# Dark Matter

## Marco Cirelli

*Laboratoire de Physique Théorique et Hautes Énergies (LPTHE),*
*CNRS & Sorbonne Université, 4 Place Jussieu, Paris, France*

## Alessandro Strumia

*Dipartimento di Fisica dell'Università di Pisa, Italia*

## Jure Zupan

*Department of Physics, University of Cincinnati, Cincinnati, Ohio 45221,USA*

## Abstract

We review observational, experimental and theoretical results related to Dark Matter.

Version 0, November 11, 2011 – Version 1, June 3, 2024 – Version 2, July 31, 2024 – Version 3, May 8, 2025

# Introduction

The era of geographical explorations of the Earth ended in the past century. Mount Everest was climbed in 1953, the Mariana Trench descended in 1960 and the North and South Poles reached around 1908 and in 1911, respectively. The past century will be remembered for an even more important milestone: we understood the structure of everyday matter, and in particular quantum mechanics and relativity have taught us that often the 'stuff' that we observe has a intrinsic nature that differs from our intuition. Does that mean that the age of scientific explorations of matter is over? Not yet. There is more matter left to be understood. In fact, the bulk of the matter in the Universe is not yet understood and goes by the name of Dark Matter (DM).

In a similar way as the geographical explorations have turned from Earth to space, so have our explorations of matter. The existence of DM is established by observations from galactic to cosmological scales. However, these observations only probe the gravitational coupling of DM — the total mass and its spatial distributions. In order to understand what DM truly is, we also need to observe its other possible interactions with ordinary matter.

Given the importance of the problem it is not surprising that the subject of DM has attracted a lot of attention: many experiments are being performed (see fig. 4.7), and presently about 10% of the papers in cosmology, astrophysics and particle physics mention 'dark matter'. Among others, several reviews about DM, with different scopes and thrusts, have been written in the past [1].

This brings us to the question: why another review? The reason, at least from our standpoint, is that the field has experienced a rapid evolution in the past years, and is now at a turning point. For example, not so many years ago the term 'neutralino' was often used as synonymous with 'WIMP' (Weakly Interacting Massive Particle), which in turn was used interchangeably with 'DM

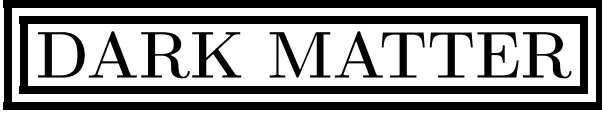

## DARK MATTER

• • • Needs confirmation • • •

## PROPERTIES

| $I(J^{PC})$ | MASS | WIDTH | DECAY MODES | PRODUCTION |
|---|---|---|---|---|
| $?(?^{??})$ | $? \pm ?$ | $? \pm ?$ | STABLE ? | $\sigma(?? \to ??) = ?$ |

Table 1: *Summary of currently known Dark Matter properties.*

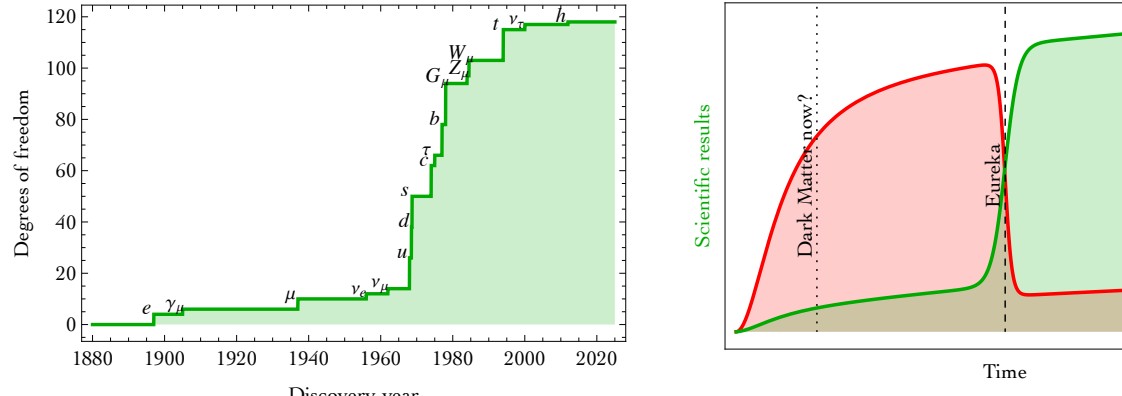

Figure 1: **Left:** *discovery year of particles.* **Right:** *typical behaviour of scientific progress. The field of DM research is nowadays ahead of its 'eureka' (discovery) moment: a vast scientific literature discusses the many open possibilities, on the basis of relatively few solid results. After that moment, hypotheses will drastically shrink and more measurements will flow.*

particle', and hence simply 'DM'. As we will discuss at length in the review, and as is obvious to those working in the field, these are drastic and narrow simplifications. They were somewhat justified by the theoretical prejudices that dominated the field of particle physics up to about a decade ago, but they are no longer justified nowadays. The rethinking of those theoretical assumptions was triggered by the fact that the very theoretical basis of the neutralinos — supersymmetry — did not show up at colliders, where experimental searches made significant progress after the start of the Large Hadron Collider, with no positive evidence so far. Furthermore, the WIMP hypothesis itself is under close scrutiny, and has had the viable parameter space reduced significantly over the past two decades. Even the once common implicit assumption that the DM is a new elementary particle is now being reconsidered.

The lack of experimental evidence for non-gravitational interactions of DM may suggest that Dark Matter is something beyond our current theoretical paradigms and the community is exploring possibilities which received less attention so far.

At this point in time it thus seems useful to review the status of DM research, from a broader point of view. This does not mean dismissing the 'traditional' research directions but rather adding new ones. The challenge one faces in doing so is that the field is fragmenting, making it difficult to decide to which topics or aspects one should give more weight in such an endeavour. The current situation in physics of DM might be similar to the point in time when ancient philosophers were discussing the nature of ordinary matter: the vague idea of atoms was proposed together with other possibilities, good ideas accumulated until the discussions stalled, waiting for the 'eureka' moment, after which a coherent picture emerged and sensible results accumulated. Table 1 summarizes the current status, and fig. 1 shows the typical learning curve of science.

Fig. 2 shows the map of recent literature: Dark Matter is at the interface between high-energy experiment, high energy phenomenology, astrophysics, and cosmology, where no paper about DM emerges as the one completely dominating the citation counts. Writing a review about DM in this time and age thus somewhat resembles the problem of summarizing a soccer match with a partial $0 - 0$ score, where one can easily fall into a trap of compiling a boring list of various attempts.

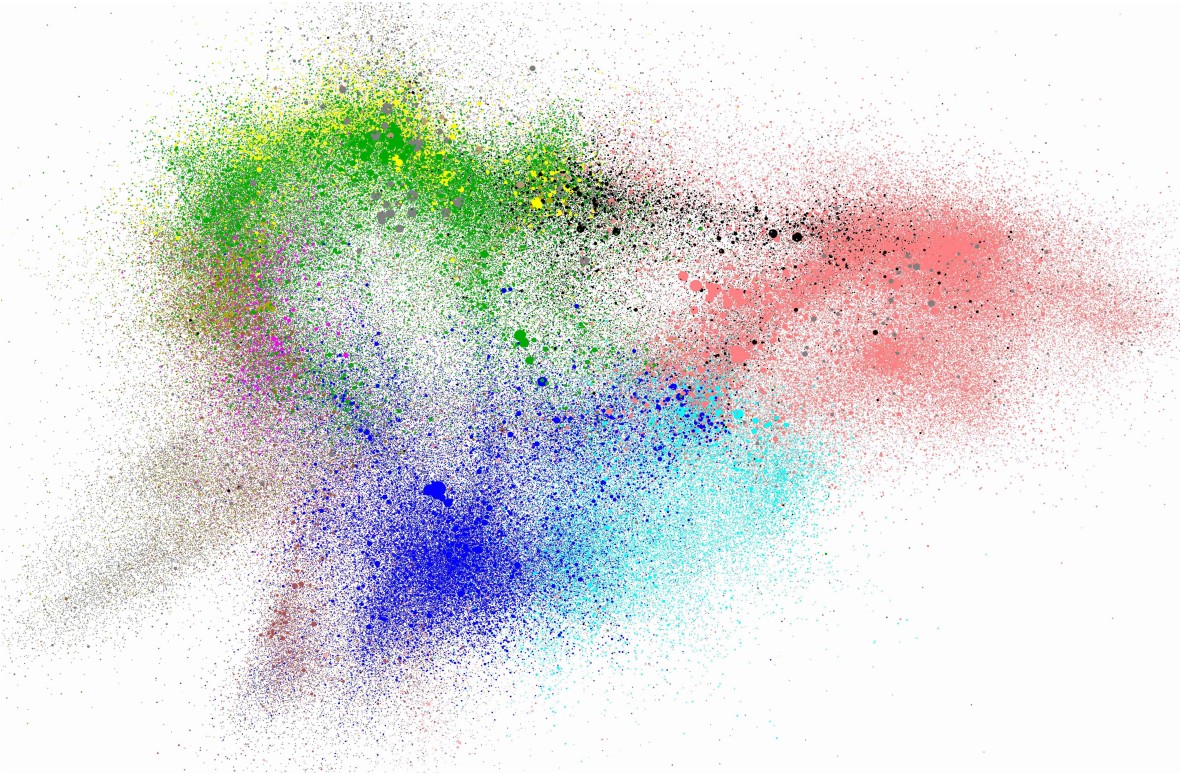

Figure 2: *Each dot is a paper, with size proportional to its number of received citations, positioned such that papers that cite each other are nearby, and colored according to its arXiv bulletin:* hep-ph, astro-ph, hep-th, gr-qc, hep-ex, nucl-th, hep-lat, *etc. Papers with 'dark matter' in the title are in black and lie at the interface between experiment, phenomenology and astrophysics. (Data from the* INSPIRE *database [2]).*

We hope we succeeded in striking instead a reasonable balance between a brief discussion of the main points of the various possibilities and a comprehensive but tedious catalogue.

This review is structured as follows:

- Part I reviews basic facts and ideas.

  - Chapter 1 summarizes evidence for DM from astrophysics and cosmology, implications for its properties, and possible alternatives.
  - Chapter 2 summarizes what we know about the DM density and velocity distribution in the Milky Way and beyond; this will be used to compute possible DM signals.
  - Chapter 3 reviews main ideas about what DM could be, from particles to waves to primordial black holes.
  - Chapter 4 summarizes ideas about how DM formed in cosmology: as a thermal relic, as a non-thermal relic, from inflation, etcetera.

- Part II deals with searches for DM.

  - Chapter 5 reviews direct detection (DM collisions with SM particles).
  - Chapter 6 reviews indirect detection (DM annihilations or decays into SM particles).

- Chapter 7 reviews collider searches (DM production from collisions of SM particles).

- Each chapter presents a summary of the present status in the different kinds of searches: DM has not been discovered yet, and limits are imposed. Then Chapter 8 reviews current anomalies that might be due to DM. While none looks compelling at present, at least they provide concrete examples of what searches mean in practice. Many expeditions failed to reach the Poles, paving the road for the ones that eventually succeeded.

- Part III reviews DM theories.

  - Chapters 9 discusses basic models of DM.

  - Chapter 10 discusses wider theoretical frameworks that include possible DM candidates: super-symmetry, extra dimensions, axions. . .

  - Alternatives to DM and their underlying theoretical frameworks are discussed in the final chapter 11.

**Acknowledgments.** We thank, for useful discussions and information over many years, Asimina Arvanitaki, Shyam Balaji, Daniele Barducci, Zurab Berezhiani, Denis Bernard, Aryaman Bhutani, Michael Blanton, Albert Bosma, Mathieu Boudaud, Giorgio Busoni, Sandrine Codis, Paschal Coyle, Davide D'Angelo, Pedro De La Torre Luque, Aloïse Dijoux, Caterina Doglioni, Fiorenza Donato, David Dunsky, Roberto Franceschini, Davide Franco, Raghuveer Garani, Yoann Génolini, Gerry Gilmore, Gaëlle Giesen, Andreas Goudelis, Benjamin Grinstein, Christian Gross, Keisuke Harigaya, Roni Harnik, Sebastian Hoof, Yonathan Kahn, Bradley Kavanagh, Jordan Koechler, Fabio Iocco, Massimiliano Lattanzi, Federico Lelli, Luka Leskovec, Luca di Luzio, Reina Maruyama, David Maurin, M. Nicola Mazziotta, Andrea Mitridate, Emmanuel Moulin, Paolo Panci, Duccio Pappadopulo, Marta Perego, Federica Petricca, Elena Pinetti, Liberato Pizza, Michele Redi, Massimo Ricotti, Dean Robinson, Nicholas Rodd, Nicola Rossi, Akash Kumar Saha, Filippo Sala, Joe Silk, Francesco Sannino, Tracy Slatyer, Marco Taoso, Diego Tonelli, Riccardo Torre, Alfredo Urbano, Natascia Vignaroli, Edoardo Vitagliano, Marta Volonteri, Andreas Weiler, Rohana Wijewardhana, Andrea Wulzer, Wei Xue, Gabrijela Zaharijas. We are also particularly grateful to all our other collaborators, since some of our recent papers arose as pieces of this review that needed original work.

We also thank, for useful additional input: Kensuke Akita, Dorian Amaral, Kazem Azizi, Torsten Bringmann, Francesca Calore, Marco Chianese, Bibhabasu De, Olivier Deligny, Mattia Di Mauro, Christopher Eckner, Andrei Egorov, Astrid Eichhorn, Matteo Fasiello, Anupam Ghosh, Sergei Gninenko, Jan Heisig, Gonzalo Herrera, Mark Hertzberg, Mudit Jain, Adil Jueid, Pyungwon Ko, Silvia Manconi, Doddy Marsh, Rahul Mehra, Maurizio Piai, Nirmal Raj, Subir Sarkar, Robert Shrock, Jing Shu, Jusak Tandean, Edoardo Vitagliano, Jenny Wagner, Tao Xu, Wen Yin, Miguel Yulo, Michael Zantedeschi.

The work of M.C. was supported in part by the *European Research Council* (Erc) under the EU Seventh Framework Programme (FP7/2007-2013) / Erc Starting Grant (agreement n. 278234 - 'NewDark' project) and by a Cnrs 80|Prime grant ('DaMeFer' project). M.C. acknowledges the hospitality of the *Institut d'Astrophysique de Paris* (Iap), the Theory Department of Cern and of the Physics Department of the University of California at Irvine (Uci) where parts of this work were done. The work by A.S. has been supported by the Neo-Nat Erc and Prin 2017fmjfmw grants. JZ acknowledges support in part by the Doe grant De-Sc1019775 and NSF OAC-2103889. JZ acknowledges support in part by the Miller Institute for Basic Research in Science, University of California Berkeley.

# Part I

# Dark Matter basics

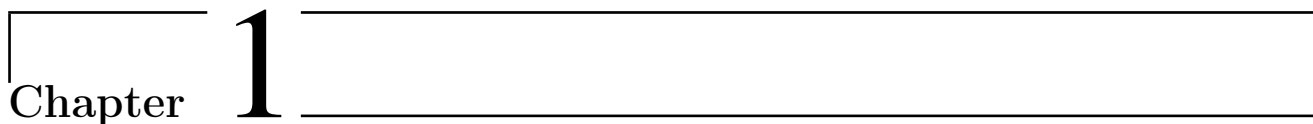

# Chapter 1

# Why? Evidence for Dark Matter

Evidence for the existence of Dark Matter (DM) comes from a wide range of astronomical scales, from a few kiloparsecs (the typical size of a small galaxy) to essentially the size of the observable Universe. It is somewhat customary to focus on three main probes: the observations of individual spiral galaxies (discussed in section 1.1), of clusters of galaxies (section 1.2), and of the Cosmic Microwave Background (CMB) and Large Scale Structures at cosmological scales (section 1.3). These observations can be consistently interpreted in terms of DM with some basic properties summarized below.[1] The three sets of probes, however, are not of equal status. The probes on the smallest scales (galaxies) are intuitive and based on simple, classical physics, though they are arguably the least useful nowadays for quantitative determinations. The probes based on the largest scales (cosmology) rely on more complex physics underlying the standard cosmological model, based on the general relativistic description of an expanding Universe. However, they do deliver the most significant evidence for DM, and are the most precise tools to restrict DM properties, as we will see below.

All these probes pertain to the *gravitational* effects of DM. No evidence based on some other DM effects (for example not due to gravitational interactions) is available yet. A natural alternative is thus to consider modifications of gravity in order to explain the phenomena usually attributed to DM. Some of the proposals in this direction are discussed in chapter 11, though none of these can explain (yet?) the full array of observations. Apart from chapter 11, we thus restrict the discussion to the standard hypothesis; that DM is made of undiscovered matter corpuscles, be they a new species of elementary particles or a new kind of macroscopic bodies.

The present cosmological DM density, averaged over the whole Universe, is usually presented in terms of the combination of parameters $\Omega_{\rm DM}h^2$, where $\Omega_i = \rho_i/\rho_{\rm cr}$ is the density relative to the critical density $\rho_{\rm cr} = 3H^2/8\pi G$, $G$ is Newton's constant and the present Hubble parameter is written as $H_0 = h \times 100\,{\rm km/s}\cdot{\rm Mpc}$, with $h = 0.674 \pm 0.005$.[2] The current best determination is [5]

$$\Omega_{\rm DM}h^2 = 0.1200 \pm 0.0012. \tag{1.1}$$

---

[1] This does not exclude the possibility that DM could be composed from a number of different substances, each of which dominates at a different scale, provided that they share the common basic properties discussed below.

[2] The combination $\Omega_{\rm DM}h^2$ is better determined from observations than $\Omega_{\rm DM}$, and in particular is not affected by the uncertainty in $h$. The quoted value of $h$ is based on Planck cosmological data [3]. As discussed in section 8.4.1, alternative determinations based on late time observations find higher values, around $h = 0.73$ [4], at odds with the value quoted above, leading to an intense debate and on doubts that uncertainties could be a factor of few higher than what reported e.g. in eq. (1.1). For definiteness, we use the cosmological value for $h$, as in [5].

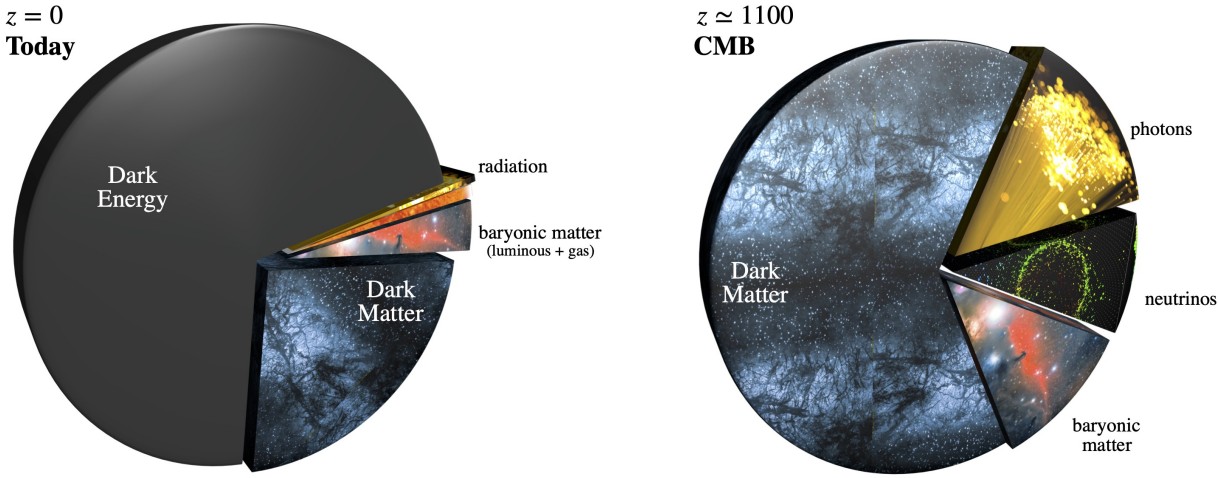

Figure 1.1: ***Content of the Universe*** *today (left) and, for comparison, at the time of photon decoupling (right). The different components evolve differently in cosmology, see appendix C.*

This trivially translates into $\Omega_{\mathrm{DM}} = 0.264 \pm 0.003$, i.e., DM constitutes about 26.4% of the total *matter-energy* content of the current Universe. Since the density of normal baryonic matter is measured to be $\Omega_{\mathrm{b}} h^2 = 0.02237 \pm 0.00015$, or 4.9%, DM constitutes about 84% of the total *matter* content (figure 1.1). Also, the current average DM density in the Universe is $\rho_{\mathrm{DM}} \approx 1.26 \ \mathrm{keV/\,cm^3}$.[3]

Next, let us summarize what the astrophysical and cosmological observations imply for the general properties of Dark Matter (in chapter 3 we will elaborate further for the specific cases, where DM is assumed to be made either of particles, fields or macroscopic bodies). The data can be reproduced assuming that DM is a kind of *cold, non-interacting, stable matter*, with *adiabatic* inhomogeneities.

- ○ **Matter** stresses the fact that DM behaves as matter in the cosmological evolution, i.e., its density decreases as the inverse of the volume. In technical terms, its equation of state parameter is $w = 0$ (see section 1.3 and appendix C.2). This is in contrast to Dark Energy, whose density, as far as we know, does not dilute in an appreciable way when the Universe expands.

- ○ **Cold** means that DM behaves as a non-relativistic fluid at the crucial time of matter-radiation equality (MReq), when structure formation begins, and henceforth during all the subsequent periods of galaxy formation. The DM corpuscles move 'slowly' and therefore attract each other and cluster. If they moved 'fast', the clustering would not be effective and structures would not form and grow. The process of clustering does re-heat the DM fluid, since its constituents gain kinetic energy after collapsing into bound structures. However, this is not enough to make DM relativistic and invalidate the general coldness of DM, except perhaps in extreme environments, such as around compact bodies (see section 6.10.4). These points will be made more quantitative in section 1.3.1 and chapter 3.

- ○ **Non-interacting** (or equivalently **collision-less**) means that the interactions of DM with itself or between DM and ordinary matter (other than the gravitational interaction, which DM certainly

---

[3]This is not to be confused with the estimated DM density at the location of the Sun in the Milky Way, see section 2.2.2. We also note in passing that some studies found the average value for DM density in the local Universe (i.e, averaged over the volume within about 10 Mpc) to be 2 to 3 times lower than in eq. (1.1) [6]. This can be explained, if by chance the Milky Way is located in an under-dense fluctuation, i.e., in a local 'void' bubble.

possesses), are small enough to be neglected. This is what differentiates DM from ordinary matter. Whereas the latter has significant interactions, including in particular the electromagnetic one, DM does not. Therein lies the reason for the naming of Dark Matter — the absence of interactions of DM with the light makes DM *dark*.

This does not mean that DM absolutely can not have any interactions (other than the gravitational one) with itself or with the ordinary matter. Quite to the contrary, in most models the production of DM in the early universe does require non-zero interactions (as discussed at length in chapter 4). Furthermore, essentially all the search strategies rely on the existence of non-zero interactions (see part II for details). Notably, DM particles with SM weak interactions, or with interactions of similar strength, fall into the 'non-interacting' class, i.e., such interactions are feeble enough to be neglected in astrophysical considerations as well as for late cosmology.

○ A corollary of non-interacting is **dissipation-less**: unlike ordinary matter, DM cannot emit electromagnetic radiation and therefore cannot easily dissipate its energy and cool down. This feature is the core reason why ordinary matter and DM behave differently during the cosmological evolution. Of course, also in this case the prohibition is not absolute: interesting models exist in which DM has some small degrees of dissipation.

○ **Stable** means that DM is present since the early phases of the Universe and has not disappeared until now (the current age of the universe being $t_{\mathrm{Uni}} \simeq 13.8$ Gyr $= 4.35 \ 10^{17}$ s). Depending on the nature of DM, this will translate quantitatively into bounds on either the decay lifetime (if DM is made of unstable particles), or the evaporation rate (if DM is made of black holes subject to Hawking radiation), et cetera. See chapter 3.

○ **Adiabatic** means that, on cosmological scales, the composition of the cosmological fluid is the same everywhere. DM has the same primordial density inhomogeneity as the other components: DM is denser where ordinary matter and photons are denser. This happens if all the inhomogeneities have been generated by a single mechanism. The mechanism is believed to be quantum fluctuations of a single inflaton field during cosmological inflation. Indeed, in agreement with this theory, all components seem to have a *Gaussian* and *quasi-scale invariant* spectrum of primordial fluctuations.

## 1.1   Mini: galaxies

### 1.1.1   Rotation curves of spiral galaxies

Spiral galaxies[4] rotate around their vertical axes. Measuring the Doppler shift of atomic lines one can determine the circular velocity of stars and other tracers (e.g., hydrogen clouds and masers) as a function of their distance from the galactic center, obtaining a *rotation curve*. Some historical and recent examples are plotted in fig. 1.2. Newton's law, $F = ma$, predicts a simple relation between the circular velocity

---

[4]Traditionally, galaxies are classified according to their observed shapes as: Elliptical (E), Spiral (S) and Barred Spiral (SB). Elliptical galaxies are further labelled according to their elongation, from E0 (round) to E7 (very elongated). They do not exhibit any particular axis of rotation and are 'dispersion supported' (as opposed to spirals and barred spirals which are 'rotation supported'). Spiral and barred spiral galaxies are further delineated according to how tight their spiral arms wrap around the center: Sa, Sb, Sc. . . and SBa, SBb, SBc. . . , in order of increasing openness of the arms. The Milky Way is a rather typical Sb galaxy. Irregular (Irr) and dwarf galaxies do not necessarily fall into one of the above classes. The classification does not necessarily correspond to an evolutionary scheme: generically, galaxies are not born as E and then evolve to S or SB (although initially it was believed to be so) nor are they guaranteed to be created as S or SB and then merge into Ellipticals (although this is a recent hypothesis). This is currently an active research domain in astrophysics.

$v_{\rm circ}$ of a test particle of mass $m$ (e.g., a star) and the mass $\mathcal{M}$ contained within a distance $r$ from the center, assuming spherical symmetry:

$$m\frac{v_{\rm circ}^2(r)}{r} = \frac{G\,m\mathcal{M}(r)}{r^2} \qquad \Rightarrow \qquad v_{\rm circ}(r) = \sqrt{\frac{G\mathcal{M}(r)}{r}}. \tag{1.2}$$

In spiral galaxies, most of the (visible) mass is concentrated in a dense central bulge and in the arms of the disk, which typically extend to $\mathcal{O}(10)$ kpcs. At large enough $r$, therefore, all the visible mass is contained within the orbit and can be replaced, in eq. (1.2), by a constant $\mathcal{M}$. The velocity should then follow a Keplerian decline $v_{\rm circ}(r) \propto r^{-1/2}$. The crucial point is that the observations of a large sample of galaxies of this kind have shown that the rotation curves instead remain *flat* (i.e., constant in $r$) out to large distances from the galactic centers. Hence, according to Newtonian gravity, additional invisible mass must be present to prevent the peripheral stars from flying away and the galaxies from breaking apart.

Assuming spherical symmetry, eq. (1.2) shows that a constant velocity $v_{\rm circ}(r)$ is obtained assuming a matter *halo* (containing a 'dark' component) that has mass density $\rho(r) \propto 1/r^2$ out to large $r$: in this way, $\mathcal{M}(r) = 4\pi \int dr'\, r'^2 \rho(r') = 4\pi\, r$ so that the dependence of $v_{\rm circ}$ on $r$ vanishes. Eventually, at even larger $r$, the halo is expected to die off and the rotation curve to start declining. However, no tracers are typically available at such large distances.

Rubin and Ford [7] performed in 1970 the first precise measurement of the rotation curve of the Andromeda galaxy (M31), tracing about 70 hydrogen clouds. They determined the curve to be rather flat out to $\sim 22$ kpc.[5] This was later corroborated by more observations, in tens of other galaxies and using radio as well as optical techniques, finding rotation curves that remain flat up to 50 kpc and beyond. The results were quickly interpreted as evidence for 'missing mass' and the problem of Dark Matter rose to prominence. To this day, with hundreds of spiral galaxies observed, the flatness of rotation curves remains probably the most intuitive and convincing evidence in favor of DM.

Realistic studies, as opposed to the simplified proof of principle sketched above, carefully model the different luminous components (bulge, bar, disk, gas...) of the observed galaxies. In astrophysical terms, it is said that they determine the proportion of invisible to visible mass, i.e., the '*mass-to-light ratio*', of a given galaxy. For a galaxy like the Milky Way or Andromeda, this ratio is of order 10. These measurements can also be used to determine the DM density profile, including in the Milky Way, although typically the result is not very constraining (see chapter 2.1.1).

## 1.1.2   Other galactic-scale evidences

There are a number of other pieces of evidence supporting the existence of DM that have been put forward, based on observations at galactic scales.

First of all, the same kinematic measurements of tracers discussed in the previous subsection can be applied to other systems, in particular to *dwarf galaxies*. These are discussed in detail in section 2.2.3. The results show evidence for a large content of DM in these systems, with a mass-to-light ratio typically on the order of 100 or more.

Moreover, the determination of rotation curves of spiral galaxies can be performed using the dwarf galaxies themselves as tracers [8]. This approach corroborates the results obtained with other methods.

In the 1970s, early numerical simulations and their analytical interpretations were showing that a dark spherical halo is needed to ensure the *stability of the disk* in spiral galaxies [9]. In its absence the

---

[5]Earlier observations do exist, notably by Babcock in 1939, Mayall in 1951 and Schwarzschild in 1954. While very interesting, they were, however, not as precise and/or did not extend as far in $r$, so that there was no consensus in their interpretations. For a detailed historical account see the review by Bertone and Hooper (2016) in [1].

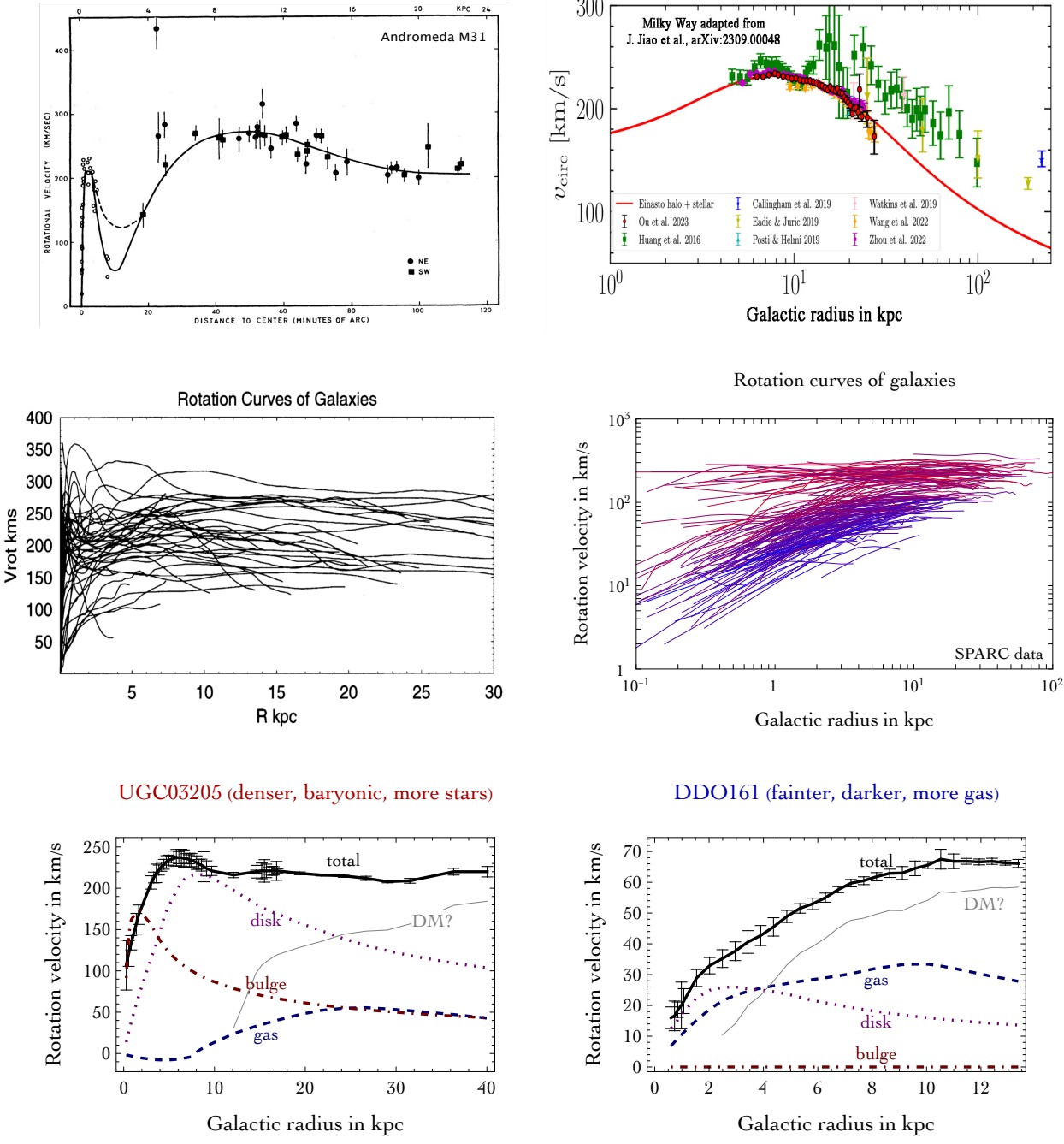

Figure 1.2: ***Rotation curves of spiral galaxies [7]***. ***Top left****: the original rotation curve of Andromeda by Rubin & Ford (1970) (©AAS, repr. with permission).* ***Top right****: a recent rotation curve for the Milky Way adapted from Y. Jiao et al. (2023) [7], that shows a hint of Keplerian decline.* ***Center left****: a compilation of about 50 galaxies, from Sofue et al. (1999) (repr. with permission).* ***Center right****: the SPARC compilation of 175 galaxies, from Lelli et al. (2016), color coded according to surface brightness (red is denser, blue is fainter).* ***Bottom left****: a denser galaxy, dominated in its center by baryons, mostly in stars.* ***Bottom right****: a fainter galaxy, dominated by a dark component, with most baryons as gas. The latter two plots show the total rotation velocity (black), the contributions from disk stars (dotted magenta), in the bulge (dot-dashed red), from gas (dashed blue), and inferred DM (grey) from eq. (2.1).*

simulated disks were seen to be disrupted within a few rotational periods and transformed into chaotic distributions of stars. However, more sophisticated works later showed that the DM halo has the negative effect of morphing a disk into a bar and eventually slowing the rotation of the bar down, which is at odds with the observed existence of many spinning disks. It seems that no simple conclusion for the role of DM can be claimed in this respect.

The existence of massive DM halos around galaxies can also be proven via **galaxy-galaxy lensing** [10]. This is the distortion of the images of background galaxies induced by the gravitational lensing effect of the foreground galaxies, analogously to how the Sun gravitationally deflects the light of background stars. The deviation of the light trajectory is $\delta\theta = 4GM_{\text{lens}}/b$, where $b$ is the impact parameter and $M_{\text{lens}}$ is the mass of the lens, in this case the foreground galaxy. Geometric optics then implies that a lens with mass $M_{\text{lens}}$ at distance $d_{\text{lens}}$ gravitationally focuses a source object (at bigger distance $d_{\text{source}}$) into an 'Einstein ring' (or arc) with angular size $\theta_{\text{E}} = \sqrt{4GM_{\text{lens}}(1/d_{\text{lens}} - 1/d_{\text{source}})}$, if the source and the lens lie on the same line of sight, and into multiple images otherwise. This phenomenon is known as *strong lensing*, despite the fact that the deviations are typically small, $\delta\theta \ll 1$, and is observable only when the lens is very massive, and the source close enough to it. More commonly observed is *weak lensing*, where the lens is not strong enough to form multiple observable images or an arc, but does lead to a distorted image of the background galaxy. By measuring the average properties, as in fig. 1.4, one can infer the amount of matter in a typical lensing galaxy, finding that it is larger than the visible mass. The same observations can also determine the typical density profile of the DM halos, found to be somewhat flattened (see section 2.4).

### 1.1.3   DM-deficient galaxies?

In 2018 van Dokkum et al. [11] claimed that the ultra-diffuse galaxy NGC1052-DF2 (close to the giant elliptical NGC1052) contains little or no Dark Matter. Its stellar mass is evaluated at $2 \times 10^8$ $M_{\odot}$, while its total mass is estimated, using the velocities of an unusual population of globular clusters as tracers, at less than a factor of 2 larger. This is a very surprising value for galaxies of this kind, which typically have a mass-to-light ratio of several hundreds. The results and the uncertainties have been closely scrutinised (e.g., whether the distance is really $\sim 20$ Mpc; if the galaxy is closer, then its inferred stellar mass would go down significantly and the mass-to-light ratio would increase). In 2019 the same group discovered NGC1052-DF4 in the same system, which had the same properties.

The interpretation of these results, if they are confirmed, and their impact on the DM paradigm, have been debated. These galaxies could be explained away as just statistical outliers or as the result of a peculiar evolution that deprived them of their DM content (e.g., tidal stripping by a massive neighboring galaxy, or mini bullet-cluster-like events, see section 1.2.1). In any case, it is fair to say that their existence poses a challenge to the standard theories of small galaxy formation. Some numerical simulations have not found any evidence of such galaxies in their results, while others have shown that small DM-deprived galaxies with the appropriate characteristics do form, thanks to close encounters with the massive host [11]. Furthermore, these results would put stress on theories that use modifications of gravity rather than DM to explain rotation curves (see section 11.3), since in these theories one should always detect a modification of gravity in presence of ordinary matter, at large enough distances.

## 1.2   Midi: clusters of galaxies

Fritz Zwicky is usually credited as having been the first person to lucidly claim the evidence for Dark Matter, in 1933. However, as discussed by G. Bertone and D. Hooper (2018) in [1], the concept and terminology of Dark Matter was already hovering in the astronomy community at the time, e.g., in the works by H. Poincaré, W. Thomson (a.k.a. Lord Kelvin), E. Öpik, J. Kapteyn, K. Lundmark and J. Oort,

as well as S. Smith. Zwicky's work is however historically important as well as particularly simple in retrospect, hence worth reviewing.

By looking at the **velocity dispersion in the Coma cluster of galaxies**, Zwicky found that extra matter was necessary to keep the galaxies together [12]. The clusters of galaxies are the largest gravitationally bound systems in the Universe. They contain hundreds to thousands of galaxies and extend to several Mpc in size (see fig. 3.1). Because of their size, galaxy clusters are good probes of the 'average' Universe. While the most precise determinations of average DM density at present do not come from galaxy clusters, they do lead to $\Omega_{\rm DM} \approx 0.2$.

Zwicky's determination was based on the virial theorem. The virial theorem links the average kinetic energy to the average potential energy, $\langle K \rangle = -\frac{1}{2}\langle V \rangle$ for gravity. In a toy system with $N \gg 1$ objects of mass $m$ at equal distance $r$ interacting through gravity, this allows to determine their total mass $mN$ from the velocity $v$ and the size $R$:

$$N\frac{mv^2}{2} = \frac{1}{2}\frac{N^2}{2}\frac{Gm^2}{R} \qquad \Rightarrow \qquad mN = \frac{2Rv^2}{G}. \tag{1.3}$$

Applying such considerations to the Coma cluster of galaxies,[6] Zwicky argued that the total mass in the cluster was larger than the visible mass, such that extra dark matter was needed.[7] The DM hypothesis was not widely accepted, but it was also not disregarded. A common interpretation was that more information would be needed in order to understand these systems.

Since the 1980s, *X*-**ray observations** have emerged as a more effective method for assessing the amount of ordinary and dark matter within galaxy clusters (for reviews see [14]). Galaxy clusters contain large amounts of ionized hydrogen and helium. When this gas collapses into the cluster's potential well, it undergoes shocks and adiabatic compression, heats up, and reaches temperatures $T_{\rm gas} \sim m_p v_{\rm esc}^2 \sim 10\,{\rm keV} \sim 10^8\,{\rm K}$, which would be typical of nuclear reactions. However, due to the low ambient density, nuclear fusion is negligible. The gas then emits *X*-rays, primarily through thermal bremsstrahlung. Assuming spherical symmetry and hydrostatic equilibrium, one can write down the equilibrium equation which links the gradient of the gas pressure, $\wp_{\rm gas}$, to the gradient of the gravitational potential, $\phi$,

$$\frac{1}{\rho_{\rm gas}(r)}\frac{d\wp_{\rm gas}}{dr} = -\frac{d\phi}{dr} = -\frac{GM(r)}{r^2}, \tag{1.4}$$

where $\rho_{\rm gas}(r)$ is the gas density and $M(r)$ the total gravitating mass (i.e., hydrogen/helium gas and dark matter) within radius $r$. Assuming an ideal gas, the pressure and density are related via $\wp_{\rm gas} = \rho_{\rm gas}kT_{\rm gas}/\mu m_p$, where $\mu \approx 0.6$ is the mean molecular weight of an admixture of $\sim 75\%$ hydrogen and $\sim 25\%$ helium. The density $\rho_{\rm gas}$ and the temperature $T_{\rm gas}$ can be measured from the intensity and the spectrum of the *X*-ray emission, thereby allowing to reconstruct the total cluster mass as well as its profile using eq. (1.4). The results confirm that the gas constitutes only a portion of the total mass of the galaxy cluster. In a way, the use of *X*-ray emissions is an application of Zwicky's method to the microscopic scale: the kinetic energy of the gas molecules (i.e., their temperature) takes the place of that of the galaxies, from which one then infers the total gravitating mass that keeps the galaxy cluster gravitationally bound.

*X*-ray measurements also prove very useful in a different way, namely in analyzing unique spatial configurations such as the collisions of galaxy clusters, which we discuss next.

---

[6]Zwicky approximated the cluster with a different toy model than in eq. (1.3): a sphere of constant density $\rho$ and radius $R$. The results differ by order one factors, with the average potential energy now given by $-\langle V \rangle = \int_0^R G\rho\, 4\pi r^2 dr\, M(r)/r = 3GM^2/5R$, where $M(r)$ is the mass enclosed in $r$ and $M$ is the total mass.

[7]Nowadays we measure that, in a typical cluster, the mass of the stars in galaxies represents $1 - 2\%$ of the total mass and the intergalactic gas represents $5 - 15\%$. The rest is missing and is interpreted as DM [13].

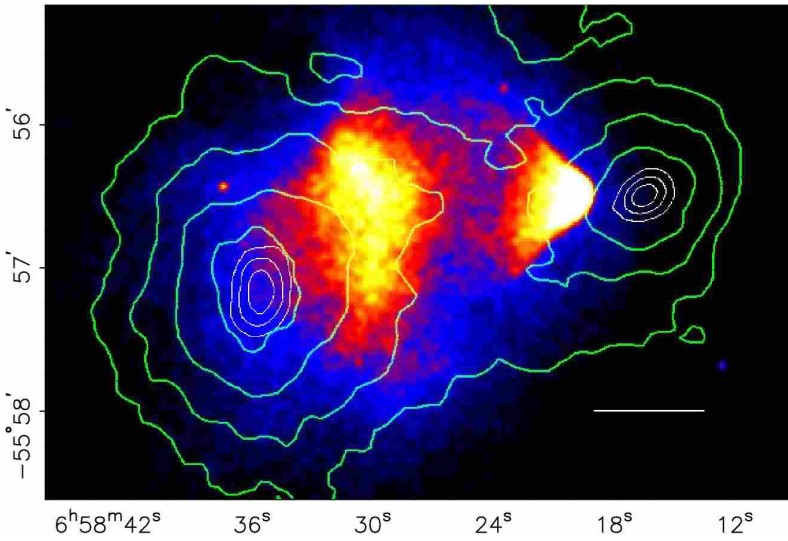

Figure 1.3:  **Bullet cluster**. *The collision of a pair of clusters of galaxies, with the colored map representing the X-ray image of the hot baryonic gas. This is displaced from the distribution of the total mass reconstructed through weak lensing, shown with green contours. The white bar corresponds to the length of* 200 kpc. *From Clowe et al. (2006) in [13] (©AAS, reproduced with permission).*

## 1.2.1  Weak gravitational lensing: the bullet cluster and cosmic shear

Today, one of the most striking evidences for the presence of DM on the length scales of galaxy clusters comes from the observations of a pair of colliding clusters known as the *bullet cluster* located 3.7 Gyr away, with a catalog name 1E0657-558 (or 1E0657-56) and first observed in detail in 2006, as well as from similar systems [13]. Most of the baryonic mass in the bullet cluster is in the form of hot gas whose distribution can be traced through its X-ray emissions. The distribution of the total mass, visible and dark, was independently measured through weak lensing.

The special feature of the bullet cluster system is that the visible matter and Dark Matter are spatially separated, see fig. 1.3. The interpretation is the following: in the past, each of the two clusters of galaxies was an ordinary system, with the visible matter and DM mixed together. The two objects collided 150 million years ago. Visible matter interacts significantly with itself, so that the hot gas from the two clusters experienced a collisional shock wave. DM, on the other hand, experienced negligible collisions with itself and with normal matter, such that the DM clouds of the two systems simply passed through each other. This led to the present separation of the visible and dark matter components, apparent in fig. 1.3.[8] After the observation of the bullet cluster, many similar systems have been studied. Harvey et al. (2015) [13] report the results on 72 of them and conclude that the existence of DM can be established with a significance of more than $7\sigma$.

This kind of observations puts a severe strain on alternative interpretations where DM is replaced by a modification of gravity. Such modifications cannot get spatially separated from normal matter (unless they too introduce something that effectively behaves as DM), so that the anomalous lensing signal would follow the distribution of the visible matter, as we will discuss in more detail in chapter 11.

---

[8]Detailed studies reconstruct an initial relative velocity of about 3000 km/s before the collision between the two clusters. This had been claimed to be unusually high: according to the tails of velocity distributions in ΛCDM cosmology, the probability of observing such an event had been claimed to be too low ($\sim 10^{-5}$ assuming a reasonable amount of matter inhomogeneities) [15]. Hence the bullet cluster, in this specific aspect of the relative speed, had been used as evidence *against* Dark Matter. Later studies, however, have disputed the claim and found probabilities that are in agreement with ΛCDM cosmology [15].

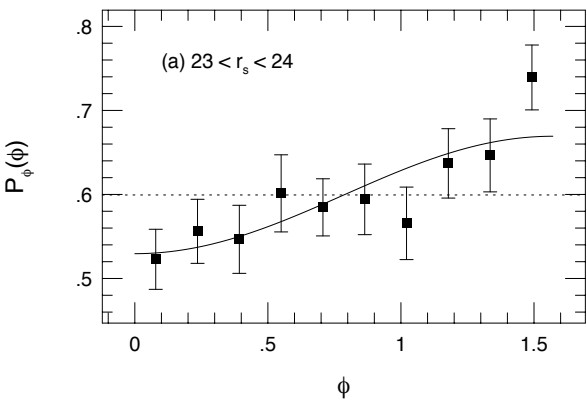
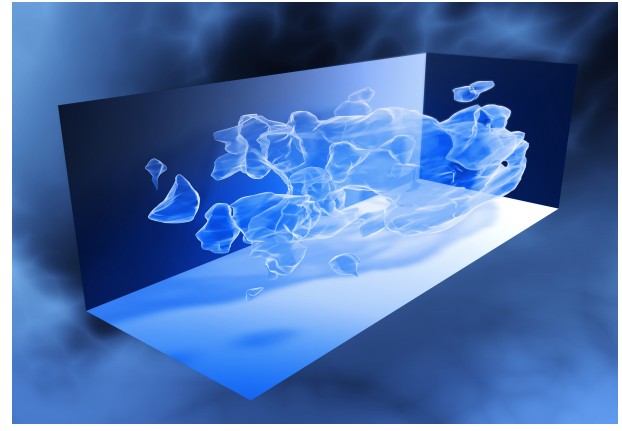

Figure 1.4:   ***Weak gravitational lensing*** *is a multi-length and multi-purpose tool to establish the existence of Dark Matter and determine its properties. At small scales the important effect is galaxy-galaxy lensing (discussed in section 1.1.2), see the left panel. The images of background galaxies are statistically distorted to align on a ring pattern around a foreground galaxy. That is, for a certain bin in magnitudes ($r_s$) of the source galaxies, the average probability of the orientation is measured to have a deficit of images oriented radially ($\phi = 0$) and an excess of images oriented tangentially ($\phi = \pi/2$). At very large scales the important effect is cosmic shear (section 1.2.1), see the right panel. The clouds in the graph are the constant density contours of the large scale DM formations, as reconstructed in 3D by the observations of the lensing of background sources. (Figures from Brainerd et al. 1995 in [10] (©AAS, reproduced with permission), credit: NASA/ESA/Richard Massey).*

The bullet cluster and similar systems also provide information about the particle physics properties of DM. The fact that the DM halos did not slow down compared to the collision-less trajectories implies an upper bound $\wp \sim \sigma\, n\, r_{\mathrm{cl}} \lesssim 1$ on the probability $\wp$ that a DM particle scatters with another DM particle on the typical scale of the system. Here $\sigma$ is the DM self-interaction cross section, $n$ the DM number density and $r_{\mathrm{cl}}$ the size of the cluster. In the specific case of the bullet cluster, data indicate $r_{\mathrm{cl}} \sim 150$ kpc and $\rho \sim 0.5\,\mathrm{GeV/cm^3}$. Observations provide information on the DM density $\rho = Mn$ (and not directly on the DM number density $n$), so one obtains a bound on $\sigma/M$, where $M$ is the DM mass. Different studies find slightly different values [13, 16], in the ballpark of (within roughly a factor of 2)

$$\frac{\sigma}{M} \lesssim 1\,\frac{\mathrm{cm^2}}{\mathrm{g}} = \frac{1.8\,\mathrm{mb}}{\mathrm{GeV}} = \frac{4580}{\mathrm{GeV^3}}, \tag{1.5}$$

when assuming point interactions. For comparison, 50 mb is a typical QCD cross section, for, e.g. $pp$ scattering. The bound in eq. (1.5) is thus satisfied if DM is made of particles somewhat heavier than protons and/or somewhat less strongly interacting than protons. The DM/matter cross section is similarly bound by astrophysics and cosmology to be roughly smaller than QCD size at $M \sim 1$ GeV, although a number of other much more stringent constraints apply (see the full discussion in section 5.1.2).

## Cosmic shear

At length scales somewhat intermediate between the galactic/cluster and the cosmic scales, the detection of *cosmic shear* also provides evidence for DM [17]. Cosmic shear refers to the deflection of light from very distant galaxies by the gravitational attraction due to the foreground mass concentrations. These are not in the form of the DM halos of galaxies or clusters (as was the case for the galaxy-galaxy lensing and for the case of cluster collisions, discussed above), but rather in the form of much larger and diffuse

structures such as vast filaments and loose clumps. The measurements are performed statistically on a very large number (millions) of distant galaxies using multi-wavelength surveys, which have to be very deep (detecting very distant galaxies) and very wide (exploring large portions of the sky). The shapes of distant galaxies are distorted by the shear, changing the major-to-minor axis ratios by about 2%, which can be detected on a statistical basis. The observations determine $\Omega_{\mathrm{DM}} \approx 0.25$.

The importance of cosmic shear goes well beyond the mere additional evidence for the existence of DM. By reconstructing the matter distribution along the line of sight, one is effectively looking back at the formation history of the large DM structures that provide the scaffolding of the Universe, which we discuss in the next section.

A phenomenon along similar lines is the so called **CMB lensing**, which involves searching for distortions of the CMB map caused by the foreground mass concentrations. In a way, this is the ultimate application of weak gravitational lensing as a tool to probe DM. Since the CMB is the earliest observable light, it provides a unique opportunity to investigate the extremely large DM structures at high redshifts ($z \gtrsim 2$). CMB lensing has been detected with high statistical significance, both in data from PLANCK as well as from other experiments (see [18] for a review and recent PLANCK results), corroborating the evidence for DM. The effect of lensing is to reshuffle the CMB perturbations. Notably, it smooths out the peaks at large $\ell$ in the CMB power spectrum and affects the polarization of the CMB photons, creating the so called $B$ modes out of $E$ modes. In fact, from the perspective of someone interested in the primary CMB, lensing is a contaminant and has to be subtracted (i.e., the CMB spectrum has to be 'delensed').

## 1.3 Maxi: the Universe

The most convincing and precise evidence for the existence of Dark Matter comes nowadays from the largest scales possible, the entire observable Universe. The basic point, qualitatively, is that the Universe would not have acquired the appearance that we observe today if not for the role that DM played: galaxies would not be distributed in the way they are, and the CMB temperature anisotropies would not look the way they do, if it weren't for DM. DM acts as an indispensable catalyser for the formation of structures. It brings the Universe from the initial state characterized by almost perfect smoothness with only tiny inhomogeneities to a state rich with structures at many different scales. It allows primordial inhomogeneities to grow, forming the galaxies observed today. As we will see later, ordinary matter cannot accomplish this task, essentially because of its coupling to the radiation.

For the discussion below, we need some aspects of the standard cosmological model. A quantitative description is given in Appendix C, with fig. C.1 summarizing the evolution of the three main components of the Universe. Qualitatively, the evolution of these three components can be briefly summarised as:

▷ **Radiation** has density that scales as $1/a^4$. Initially, when the scale factor $a$ was very small, radiation (photons and neutrinos) dominated, and the Universe was almost homogeneous and opaque to photons.

▷ **Matter** has density that scales as $1/a^3$. At later stages the Universe became matter dominated (at $t \approx 50\,\mathrm{kyr}$, when the Universe was $a_{\mathrm{eq}} \approx 1/3400$ smaller than today) and then transparent (the 'recombination' happened at $t \approx 380\,\mathrm{kyr}$, i.e., at $a_{\mathrm{recomb}} \approx 1/1100$, when the Universe cooled well below the hydrogen binding energy so that electrons and protons formed neutral hydrogen).

▷ **Dark energy**. Now (scale factor $a = 1$) the Universe is dominated by a component, whose density does not evolve with $a$, compatible with it being a cosmological constant or vacuum energy.

In the remainder of this chapter we work inside the framework of such a standard cosmological model. In the next subsections we compute the formation of Large Scale Structures (LSS), while in subsection 1.3.3 we discuss the imprint of DM on the acoustic peaks of the CMB. These two probes are illustrated in fig. 1.5.

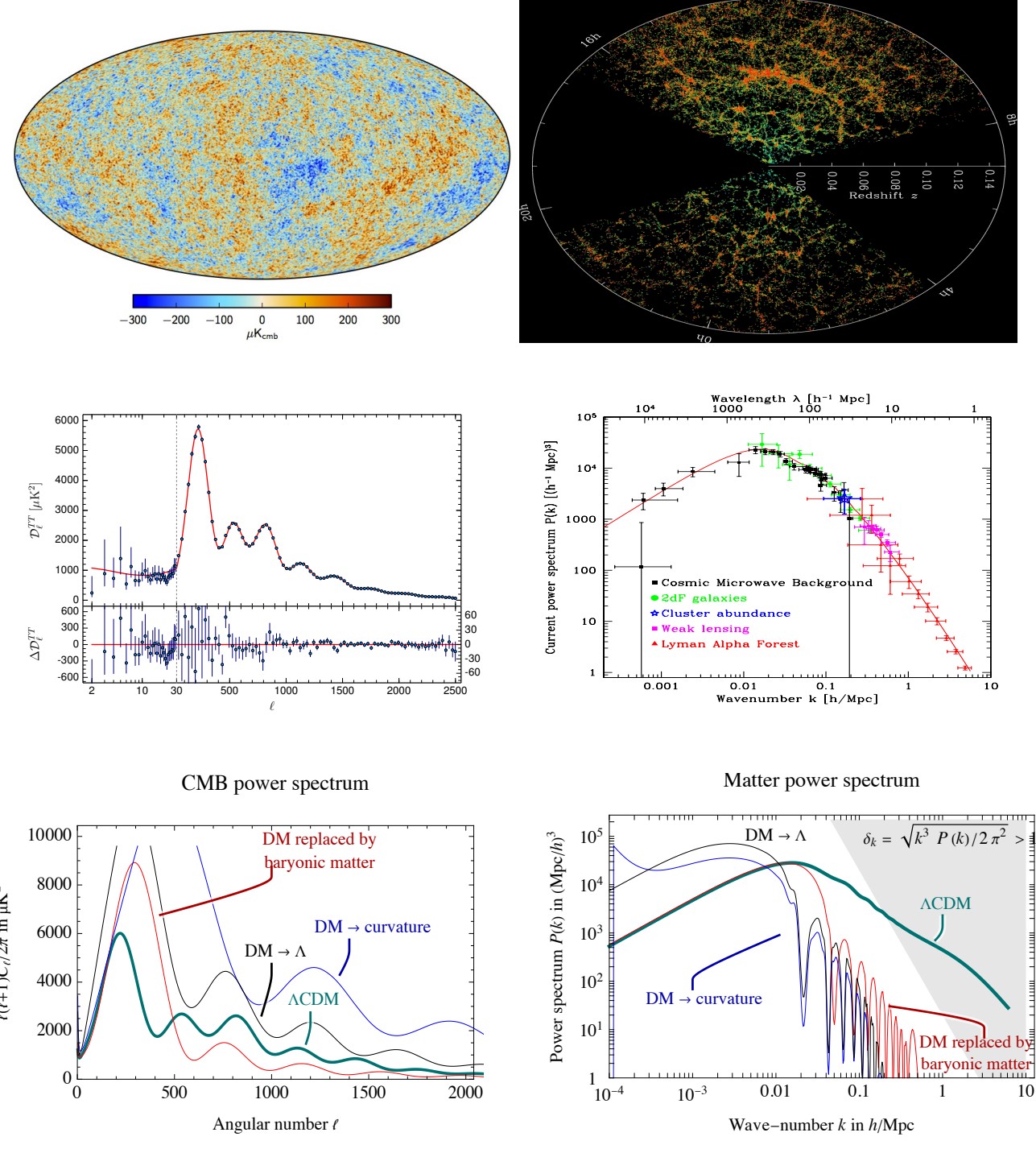

Figure 1.5:   *The power spectrum of the **CMB acoustic peaks** (middle row, left) is extracted from the map of **temperature anisotropies** (top left). The **matter power spectrum** (middle row, right) is extracted from extensive **galaxy surveys** (top right) as well as from other mapping probes. The same quantities in absence of DM are illustrated in the bottom row. Figures: from [3, 19], ©ESO, ©APS, reproduced with permission; courtesy of M. Blanton and SDSS; created with the* CAMB *web interface.*

## 1.3.1 Large scale structure formation

Today, the Universe is very inhomogeneous on scales smaller than its present horizon. Take for instance the results of galaxy surveys such as the one shown in fig. 1.5 (top right), which corresponds to a 3-dimensional map of a good fraction of all galaxies in the observable Universe. Already by eye one can resolve a variety of structures: lumps, filaments, walls, voids, which appear at different scales. Other astronomical observations, such as the Lyman-$\alpha$ forest, weak lensing measurements, cluster counts, etc, concur in establishing the picture of a clumpy Universe. Simulations and observations suggest that voids are regions with density $\sim 10\times$ smaller than the average: they fill $\sim 77\%$ of the volume, while containing only $\sim 15\%$ of the total mass [20]. Walls occupy about 18% of the volume, while containing about a quarter of the mass. Filaments provide the dominant contribution to gravitational forces: they occupy about 6% of the volume, and contain about half of the mass. Lumps occupy a small sub-percent fraction of the volume, and contain $\sim 11\%$ of the mass.

Quantitatively, from all these measurements one can extract the *matter power spectrum $P(k)$* plotted in fig. 1.5 (middle row on the right). It is defined via $\langle \delta_k \delta_{k'} \rangle = (2\pi)^3 \, P(k) \, \delta^3(\boldsymbol{k} - \boldsymbol{k}')$, where $\delta_k$ is the Fourier transform of the density contrast $\delta(\boldsymbol{r})$, defined more precisely in eq. (1.12) below, and $\langle \, \rangle$ denotes an average over $\boldsymbol{k}$ orientations. The Dirac delta $\delta^3(\boldsymbol{k} - \boldsymbol{k}')$, not to be confused with the $\delta$ that denotes the perturbations, means that modes with different $\boldsymbol{k} \neq \boldsymbol{k}'$ happen to be statistically independent. This property can be understood from inflation, a theory that predicts average statistical properties encoded in $P(k)$. Since $P(k)$ is the Fourier transform of the density self-correlation function, it conveniently expresses in Fourier space the inhomogeneity in matter distribution in the physical space. A large (small) value of $P(k)$ means that there exist many (few) structures with the characteristic size $\sim 1/k$. The measurements shown in fig. 1.5 (middle right) therefore show that the Universe has 'some power on all scales'. To make the clumpiness of the Universe even more apparent, sometimes $P(k)$, which has units of (length)$^3$, is rescaled into the dimension-less variance $\Delta^2(k) = k^3 P(k)/2\pi^2$. Small values of $\Delta^2$ correspond to small density contrasts, while $\Delta^2 \sim 1$, for instance, indicates an over-density comparable to the average. A quick manipulation of the $P(k)$ data in fig. 1.5 (middle right) shows that indeed $\Delta^2$ rises to large values in the large $k$ region. In other words, the data show that the Universe exhibits large inhomogeneities on small scales (large $k$).

In the standard cosmological model the primordial inhomogeneities are generated from inflation with small amplitudes, $\delta \approx 10^{-5}$. This is confirmed from the near-perfect smoothness observed in the CMB, which essentially provides a photo of the photon content of the Universe, taken when photons last scattered.

This poses a question: how could the tiny primordial inhomogeneities grow from such small amplitudes to the large contrasts that we observe today? The answer, as anticipated above, is that the growth of the density perturbations is crucially driven by the presence of DM. In order to quantitatively understand how that happened, we need to sketch their evolution in the early Universe.[9]

Let us consider a Universe filled with a generic matter fluid, whose exact nature we leave for the moment unidentified. The main features of the evolution of density perturbations of such a medium can be understood by approximating the full general-relativistic computation with its Newtonian limit. This is a valid description of non-relativistic matter at length scales smaller than the horizon. A non-relativistic fluid is fully characterized by its density, $\rho(\boldsymbol{r}, t)$, by its velocity field, $\boldsymbol{v}(\boldsymbol{r}, t)$ and by the equation of state for the pressure $\wp(\rho)$. Its gravitational interactions are described by the Newton potential $\Phi$. These

---

[9]Cosmological perturbation theory and structure formation is a whole research field in itself. Here, we focus only on the aspects that are instrumental to our goal, which is to show the crucial role that DM plays in the growth of such structures. We follow quite closely the treatment found in classic textbooks [21].

quantities are governed by the evolution ('Euler') equations

$$
\begin{cases}
\dfrac{\partial \rho}{\partial t} + \boldsymbol{\nabla} \cdot (\rho \boldsymbol{v}) = 0, & \text{continuity,} \\[2mm]
\dfrac{\partial \boldsymbol{v}}{\partial t} + (\boldsymbol{v} \cdot \boldsymbol{\nabla})\boldsymbol{v} = -\dfrac{\boldsymbol{\nabla}\wp}{\rho} - \boldsymbol{\nabla}\Phi, & \text{Newton law } \boldsymbol{a} = \boldsymbol{F}/m, \\[2mm]
\nabla^2 \Phi = 4\pi G \rho, & \text{Poisson,}
\end{cases}
\tag{1.6}
$$

which form a system of non-linear differential equations. For a quasi-homogeneous universe it is possible to gain useful analytic information by expanding the above quantities as the 1st-order perturbations to a 0th-order homogeneous background

$$
\rho = \rho_0(t) + \rho_1(\boldsymbol{x}, t), \qquad \wp = \wp_0 + \wp_1, \qquad \boldsymbol{v} = \boldsymbol{v}_0 + \boldsymbol{v}_1, \qquad \Phi = \Phi_0 + \Phi_1.
\tag{1.7}
$$

We first consider the case of a static universe (no expansion). Eq.s (1.6) reduce to a set of coupled equations for the perturbations

$$
\begin{cases}
\dfrac{\partial \rho_1}{\partial t} + \rho_0 \boldsymbol{\nabla} \cdot \boldsymbol{v}_1 = 0, \\[2mm]
\dfrac{\partial \boldsymbol{v}_1}{\partial t} + \dfrac{v_s^2}{\rho_0}\boldsymbol{\nabla}\rho_1 + \boldsymbol{\nabla}\Phi_1 = 0, & \text{(static Universe)} \\[2mm]
\nabla^2 \Phi_1 = 4\pi G \, \rho_1,
\end{cases}
\tag{1.8}
$$

where we defined the quantity $v_s^2 = \partial\wp/\partial\rho = \wp_1/\rho_1$ which is interpreted as the sound speed in the fluid (as we will see in a moment). By acting with $\partial/\partial t$ on the first equation, and performing appropriate substitutions in it using the second and third equations, one arrives at one evolution equation for the density perturbation $\rho_1$, i.e., at the *Jeans equation*,

$$
\frac{\partial^2 \rho_1}{\partial t^2} - v_s^2 \nabla^2 \rho_1 = 4\pi G \, \rho_0 \, \rho_1.
\tag{1.9}
$$

Ignoring gravity (i.e., setting $G = 0$), the solution are density waves (i.e., sound) that travel at speed $v_s$. Including gravity, the full equation expresses the competition between a pressure term (on the left-hand side) and a collapse term (on the right). The Jeans length $\lambda_J = \sqrt{v_s^2/(4\pi G \rho_0)}$ discriminates which term is dominant: perturbations on large scales $\lambda > \lambda_J$ collapse and grow in time, while perturbations on small scales $\lambda < \lambda_J$ are supported by pressure. The growth with time of the collapsed modes is exponential, $\rho_1 \propto e^{\sqrt{4\pi G \rho_0}\, t}$, as one can easily check by solving the differential equation (1.9) for $v_s \to 0$. This is the Jeans instability that, when applied to normal matter, explains how gas clouds collapse to form compact bodies, e.g., stars. The intuitive meaning of the Jeans scale is that a cloud of gas collapses when it is so big that its hydrostatic pressure, which prevents the collapse on time-scales $\tau_{\text{pressure}} \sim \lambda_J/v_s$, is too slow to stop the gravitational attraction, which clumps it on a time-scale $\tau_{\text{gravity}} \sim (G\rho_0)^{-1/2}$. One can also define the Jeans mass as $M_J = 4\pi\rho_0\lambda_J^3/3$, i.e., as the mass enclosed in a sphere of radius $\lambda_J$. Perturbations with mass $M > M_J$ are 'Jeans unstable' and collapse.

We move next to the case of an expanding universe. Solving the 0th-order quantities describing the smooth background gives the Hubble expansion

$$
\rho_0(t) = \frac{\rho_0(t_0)}{a^3}, \qquad \boldsymbol{v}_0 = H\boldsymbol{r}, \qquad \boldsymbol{\nabla}\Phi_0 = \frac{4\pi G \rho_0}{3}\boldsymbol{r}.
\tag{1.10}
$$

So far, $a(t)$, the usual scale factor of the Universe, has been left with an unspecified time dependence. Solving eq.s (1.6) would determine the expansion $a(t)$ of a matter-dominated universe. In general, extra components (radiation and the cosmological constant) contribute to the expansion, and $a(t)$ is given by

the fully relativistic Friedmann equations, see Appendix C. The first relation in (1.10) is just the standard dilution of non-relativistic matter when the volume expands by a factor of $a^3$. The second relation is the Hubble law for the velocity field of the homogeneous background, with respect to which the perturbations $\boldsymbol{v}_1$ can be seen as peculiar velocities. The third is the solution to the Poisson equation, applying the Gauss theorem to a sphere with radius $r$ centered at the origin and ignoring the infinite amount of outside matter; this logical jump, known as the 'Jeans swindle', is justified thanks to the existence of a horizon in the full relativistic treatment.

Expanding eq. (1.6) to 1st-order in the perturbations leads, after tedious but straightforward derivations, to the linear equations

$$
\begin{cases}
\dfrac{\partial \rho_1}{\partial t} + 3H\rho_1 + H(\boldsymbol{r} \cdot \boldsymbol{\nabla})\rho_1 + \rho_0 \boldsymbol{\nabla} \cdot \boldsymbol{v}_1 = 0, \\[2mm]
\dfrac{\partial \boldsymbol{v}_1}{\partial t} + H\boldsymbol{v}_1 + H(\boldsymbol{r} \cdot \boldsymbol{\nabla})\boldsymbol{v}_1 + \dfrac{v_s^2}{\rho_0}\boldsymbol{\nabla}\rho_1 + \boldsymbol{\nabla}\Phi_1 = 0, \qquad \text{(expanding Universe).} \\[2mm]
\nabla^2 \Phi_1 = 4\pi G \rho_1,
\end{cases}
\tag{1.11}
$$

The extra terms in eq. (1.11), as compared to eq. (1.8), are identified by the appearance of $H$. Next, we define the relative density (or density contrast) $\delta(\boldsymbol{r})$ and expand it in co-moving Fourier modes:

$$
\delta(\boldsymbol{r}) \equiv \frac{\rho_1(\boldsymbol{r})}{\rho_0} = \int \frac{d^3 k}{(2\pi)^3}\, \delta_k(t) \exp\left[-i\boldsymbol{k}\cdot\boldsymbol{x}\right], \qquad \text{where} \quad \boldsymbol{x} \equiv \frac{\boldsymbol{r}}{a(t)}.
\tag{1.12}
$$

The rescaling of the space vector by a factor $a(t)$ means that the wavenumber $1/k$ follows the average evolution of the universe. This is convenient because, in this way, equations for modes $\delta_k$ with different $k$ turn out to be decoupled. Similarly, we define $\boldsymbol{v}_k$ and $\Phi_k$, the Fourier transforms of the velocity perturbation $\boldsymbol{v}_1$ and of the gravitation potential $\Phi_1$. In Fourier space eq. (1.11) becomes

$$
\begin{cases}
\dfrac{\partial \delta_k}{\partial t} - \dfrac{i\boldsymbol{k}}{a} \cdot \boldsymbol{v}_k = 0, \\[2mm]
\dfrac{\partial (a\,\boldsymbol{v}_k)}{\partial t} - i\boldsymbol{k}v_s^2 \delta_k - i\boldsymbol{k}\Phi_k = 0, \\[2mm]
\Phi_k = -\dfrac{4\pi G\rho_0}{k^2} a^2\, \delta_k.
\end{cases}
\tag{1.13}
$$

The second equation in (1.13) can be simplified further. One can decompose the velocity perturbation as $\boldsymbol{v}_1 = \boldsymbol{v}_{1\perp} + \boldsymbol{v}_{1\parallel}$, where $\boldsymbol{\nabla} \cdot \boldsymbol{v}_{1\perp} = 0$ (divergence-free or solenoidal component) and $\boldsymbol{\nabla} \times \boldsymbol{v}_{1\parallel} = 0$ (curl-free or irrotational component). In Fourier space $\boldsymbol{k} \cdot \boldsymbol{v}_{k\perp} = 0$, $\boldsymbol{k} \times \boldsymbol{v}_{k\parallel} = 0$. The second equation in (1.13) for $\boldsymbol{v}_{k\perp}$ just amounts to $\partial(a\,\boldsymbol{v}_{k\perp})/\partial t = 0$, giving $\boldsymbol{v}_{k\perp} \propto 1/a$. The solenoidal component $\boldsymbol{v}_{k\perp}$ therefore dies away with the expansion of the Universe, and only the irrotational component, $\boldsymbol{v}_{k\parallel}$, survives. Since $\boldsymbol{v}_{k\parallel}$ is parallel to $\boldsymbol{k}$, we can substitute everywhere in eq. (1.13) $\boldsymbol{v}_k \to v_k \hat{\boldsymbol{k}}$ (where $\hat{\boldsymbol{k}}$ is the unit vector along $\boldsymbol{k}$) and $\boldsymbol{k} \to |\boldsymbol{k}| \equiv k$. At this point we can combine the three equations (1.13) into one second order equation. By taking $v_k$ from the first equation, plugging it into the second and using $\Phi_k$ from the third, we arrive at

$$
\frac{\partial^2 \delta_k}{\partial t^2} + 2H\frac{\partial \delta_k}{\partial t} + \left(\frac{v_s^2 k^2}{a^2} - 4\pi G\rho_0\right)\delta_k = 0.
\tag{1.14}
$$

This is the Jeans equation for density perturbations, analogous to eq. (1.9) but in the Fourier space and for a Universe that is expanding with Hubble parameter $H$. Similarly to the static case, the pressure term (proportional to $v_s^2$) wins over the collapse term (proportional to $G\rho_0$) for small length scale modes, i.e., for $k > k_J \equiv a\sqrt{4\pi G\,\rho_0/v_s^2}$. The critical Jeans wavenumber $k_J$ now depends on $a$, indicating that different scales start clustering at different times. Furthermore, the expansion of the Universe has a friction-like effect (the extra term in (1.14) proportional to $H$) and thereby slows the clustering of matter.

It is now time to specify which kind of matter fluid we are dealing with, and in which epoch of the Universe's evolution it is relevant.

**Non-clustering of baryonic matter.** Before considering Dark Matter, let us first focus on the case of baryonic matter coupled to photons. This means that the protons (as well as nuclei) and the electrons, which in cosmology are collectively called the 'baryonic matter', are *tightly coupled* by electromagnetism to the photons. They form a matter fluid with the relativistic sound speed $v_s \approx c/\sqrt{3}$, where $c = 1$ is the speed of light. With this extra input from relativistic physics, we can then proceed with the non-relativistic Jeans equation for density perturbations, eq. (1.14). The large speed of sound, $v_s \sim \mathcal{O}(1)$, means that the large pressure provided by photons dominates over the gravitational term in eq. (1.14). One thus expects that such a fluid does not cluster. This is confirmed by the explicit solutions of eq. (1.14). In a matter-dominated universe, i.e., in the case most favourable for clustering since radiation does not cluster, one has $\rho_0 = 3H^2/8\pi G$, $a(t) = (\frac{3}{2}tH_0)^{2/3}$ and $H = \dot{a}/a = 2/3t$, see Appendix C. Eq. (1.14) then becomes

$$\frac{\partial^2 \delta_k}{\partial t^2} + \frac{4}{3t}\frac{\partial \delta_k}{\partial t} + \frac{v_s^2 k^2}{\left(\frac{3}{2}H_0\right)^{4/3} t^{4/3}}\delta_k = 0, \qquad \text{giving} \qquad \delta_k = \frac{c_1 \cos(c_0\, kt^{1/3}) + c_2 \sin(c_0\, kt^{1/3})}{k\, t^{1/3}}. \quad (1.15)$$

For simplicity we do not specify the coefficients $c_0$, $c_1$ and $c_2$, which depend on $v_s$ and $H_0$. The main result is that this solution is *oscillating* (pressure waves, observed as peaks in the CMB power spectrum) and *damped* in time. The baryonic tightly-coupled fluid does not cluster on scales $k > k_J(a) \sim aH/v_s$, with the latter equal to the horizon, given that $v_s \sim \mathcal{O}(1)$. In other words, normal matter without Dark Matter only clusters at late times, after it decouples from photons.

**Clustering of Dark Matter.** What makes Cold Dark Matter different is that it does not interact with photons and is collectively described by a simple fluid with non-relativistic sound speed, $v_s = 0$. This makes a huge difference in cosmology. Keeping only the gravity term in eq. (1.14) and considering again the matter-dominated phase of the Universe, one obtains

$$\frac{\partial^2 \delta_k}{\partial t^2} + \frac{4}{3t}\frac{\partial \delta_k}{\partial t} - \frac{2}{3t^2}\delta_k = 0, \qquad \text{giving} \qquad \delta_k = c_{\text{grow}}\, t^{2/3} + c_{\text{decay}}\, t^{-1}. \quad (1.16)$$

This solution contains a decaying mode that dies off, and, most importantly, contains a mode that grows with a specific positive power of time. This is how the Universe evolved from a primordial quasi-homogeneous state with $\delta_k \sim 10^{-5}$ to the present clumpy state. Incidentally, note that the exponential growth found in the static case has become a power-law growth in the expanding Universe: expansion acts like a brake that slows down the clustering.

For completeness, we can also consider the DM fluid in the Radiation-Dominated (RD) epoch, described by $H = 1/2t$. In this case, one can show that eq. (1.14) loses both the $v_s^2 k^2 \delta_k/a^2$ term (because $v_s = 0$ for the DM fluid) and the $4\pi G\rho_0\delta_k$ term (because the relevant $\delta_k$ would be the one of radiation, which however does not cluster and therefore does not source perturbations in the gravitational potential). Eq. (1.14) therefore reduces to

$$\frac{d^2 \delta_k}{dt^2} + \frac{1}{t}\frac{d\delta_k}{dt} = 0, \qquad \text{giving} \qquad \delta_k \propto \ln t + \text{const.} \quad (1.17)$$

Hence, DM perturbations grow only logarithmically during radiation domination. Similar computations show that DM does not cluster efficiently when the Universe is dominated by the vacuum energy, or by curvature.

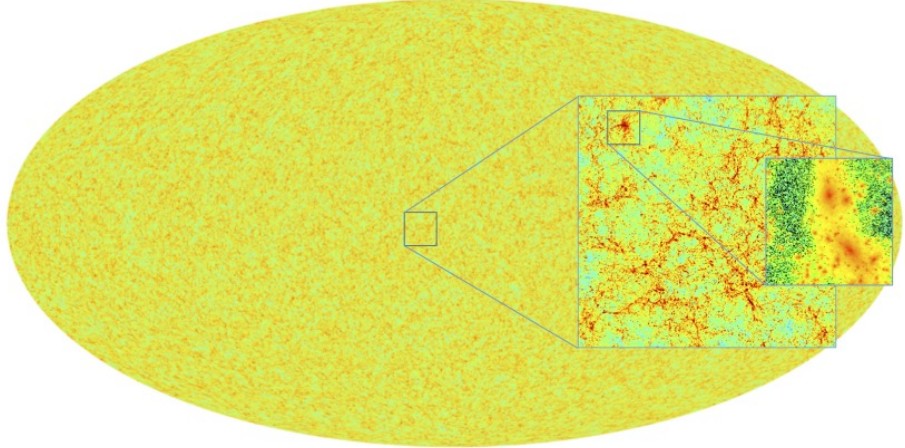

Figure 1.6:   *Results from a full-sky* **N-body DM simulation** *(from Potter et al. in [22]).*

In summary, the standard history of structure formation is as follows. The Universe became matter-dominated when the scale factors was $a_{\rm eq} \sim 1/3400$. At that point, primordial DM inhomogeneities $\delta_k \sim 10^{-5}$ began to grow as $t^{2/3} \propto a$ on length scales $1/k$ which are 3400 times smaller than the present horizon. Larger length scales started clustering later, as soon as they became smaller than the expanding horizon. During matter domination a mode $k$ re-entered the horizon when the scale factor was $a_k \approx H_0^2/k^2$. Up to $\mathcal{O}(1)$ factors, this result follows from imposing $k/a_k \approx H(a_k) = H_0\sqrt{\Omega_m/a_k^3}$. At present time we then have

$$\delta_k \approx 10^{-5}(a_0/a_k) \approx 10^{-5}(k/H_0)^2, \tag{1.18}$$

which is larger than unity on scales $H_0/k \lesssim 10^{-5/2}$, about 3 orders of magnitude smaller than the current horizon.

Normal matter, in contrast, could not cluster because it was tightly coupled by electromagnetism to photons and was forming pressure waves. Later in the evolution of the Universe, at $a_{\rm recomb} \sim 1/1100$, the temperature became low enough, and the photon bath diluted enough, that electrons and positrons could bind to form neutral hydrogen. At that point normal matter decoupled from radiation and began falling into the gravitational potential wells that DM had already started to form. In this sense, DM is a key ingredient in the construction of the 'cosmic scaffolding' observed in the Universe.

The observed matter power spectrum $P(k)$, or, equivalently, the variance $\Delta^2(k)$, corroborate this history. If the Universe contained baryonic matter only, perturbations would have had a much smaller power and would have exhibited much more pronounced oscillations (bottom right panel of fig. 1.5), in clear disagreement with the observations. A Universe containing the 'right mix' of baryons and DM, i.e. in the proportions given in eq. (1.1), agrees with data. The power spectrum changes slope at $k \gtrsim 3400\,\Omega_{\rm rad}H_0 \sim 10^{-2}/{\rm Mpc}$ corresponding to scales entering the horizon at matter/radiation equality and therefore undergoing the growth during matter domination. On top of the smooth function, the small wiggles around $k \approx 0.1\,h\,{\rm Mpc}^{-1}$, called *'baryonic acoustic oscillations'* (BAO), are the imprints of pressure waves due to the subdominant population of baryons.

As an aside, an important lesson of the above formalism is that Dark Matter made of objects with sizeable speed $v$ would not be described by $v_s = 0$ — rather, it would diffuse, suppressing the small-scale inhomogeneities. Observational data therefore constrain the DM temperature, or equivalently, the average kinetic energy of DM, to be much smaller than the DM mass. For thermalized particles, this means $M \gtrsim {\rm keV}$, as discussed in section 3.3.1.

## 1.3.2   Non-linear growth of inhomogeneities: numerical simulations

The previous subsection discussed analytically how primordial density perturbations $\delta_k$ grow due to gravitational attraction, in the perturbative limit $\delta_k \ll 1$. Over-densities eventually reach $\delta_k \sim 1$ so that the perturbative computation no longer applies. At this point gravitationally bound systems start to form: galactic subhalos, galaxies, clusters of galaxies, etc. *Smaller structures form first*, as inhomogeneities $\delta_k(t)$ are larger at smaller scales $1/k$ (see above). Larger structures form later, from accretion of diffuse DM and from merging of pre-existing smaller structures (this process is referred to as *hierarchical structure formation*).[10]   The dynamics of the growth is still described by the gravitational collision-less Boltzmann equations, written in the Eulerian form in eq. (1.6), but now the non-linear effects become relevant. The study of the evolution of inhomogeneities beyond $\delta_k \sim 1$ is therefore done via computer simulations [22]. These are a remarkable feat as they are able to cover huge ranges: $\sim 10^{13}$ years in time (from initial conditions at $z \approx 100$, $t \approx 10$ Myr to $z \approx 0$ today), $\sim 7$ orders of magnitude in space (some recent simulations deal with volumes of $\mathcal{O}(10\,\text{Gpc})^3$ with $\sim$kpc resolution), $\sim 11$ orders of magnitude in density contrast (from $\delta_k \approx 10^{-5}$ at recombination to $\delta_k \approx 10^6$, the typical density contrast in collapsed large scale structures today) and up to $\sim 30$ orders of magnitude in mass (the recent simulations by Wang et al. (2020) [22] are able to resolve at the same time Earth-mass halos, $\sim 10^{-6} M_\odot$, and cluster-mass halos, $\sim 10^{14} M_\odot$, and they can trace $10^{-11} M_\odot$ elements within a total simulated mass of $10^{19} M_\odot$, thanks to multi-zoom techniques).

What is done in practice is to simulate the gravitational motion of a large number $N$ of massive points (hence the name **$N$-body simulations**) moving in the expanding background of the Universe, starting from typical realizations of the primordial inhomogeneities. Present computers can solve Newtonian equations of motion for $N \sim 10^{12}$ massive points.[11]   While elementary DM particles are much more numerous, thanks to the universal nature of gravity this is enough to simulate properly the physics on scales bigger than the size of a single element, where the mass of each element is $\mathcal{M}/N$, with $\mathcal{M}$ the total mass in the computed volume.

At the qualitative level, such simulations produce images and movies that show how DM clusters into filaments, walls and pancakes that merge in halos separated by voids, producing what is often referred to as the *cosmic web*, see fig. 1.6. For instance, they show how objects with total mass $\approx 10^{12} M_\odot$, such as the Milky Way, mostly form at redshift $z \approx 1$, while presently about 20% of the DM remains diffuse. At the quantitative level, simulations allow to predict statistical properties of the formed structures: the distribution in mass (the halo mass function), the typical shape (spherical or ellipsoidal), the typical density profiles $\rho(r)$, the DM velocity distribution and its variance $\sigma^2(r)$. We will return to these quantities when discussing detailed results about the DM distribution in section 2.1.3 and 2.2, and possible anomalies in section 8.5. The claim here is more general: the Dark Matter hypothesis leads to an overall agreement between the qualitative and quantitative results of these simulations and the observed properties of the large scale structure in the current Universe. As hinted to several times already, DM is the scaffolding that shapes and supports the distribution of the visible matter in the Universe. Numerical simulations are instrumental in testing its presence because they provide the essential link between theoretical models for the origin and evolution of cosmological structure and the actual astronomical observations.

The discussion so far has concerned DM-only numerical simulations, performed since the 1980's. Ordinary matter (a.k.a. 'baryons') is less abundant than DM, but its effects are very important and should

---

[10]The density of forming structures is roughly given by the cosmological density at the formation time, therefore smaller structures have higher density, in agreement with fig. 3.1.

[11]Approximations based on multipoles and dedicated techniques allow the reduction of their $N^2$ gravitational forces computation to an $\mathcal{O}(N)$ numerical complexity. Furthermore, the gravitational forces are softened on small scales to avoid unphysical scatterings among nearby particles due to numerical issues. The simulations are performed assuming periodic boundary conditions.

be included, which is what recent simulations (called **baryonic** or **hydrodynamical simulations**) started to do, trying to achieve precision [23]. Discussing in detail how these simulations model baryons and their interactions goes beyond the focus of our review. In general terms, these simulations need to take into account a large number of different physical processes, on different time-scales and for very large ranges of velocities, densities, temperatures, viscosities, etc. The ingredients include: gas hydrodynamics, gas cooling, radiation, magnetic fields (since these significantly contribute to pressure; their amplification, included in the computer codes, presumably makes their more complicated seeds irrelevant), star formation and star death, as well as the dynamics of stellar ejecta (including supernovæ: the resulting cosmic rays contribute to pressure in the interstellar medium and significantly change the gravitational potential by moving matter from a localized center to a wide surrounding region within a short timescale), super-massive black holes (present in massive galaxies; their seeds are not known, they presumably grow acquiring gas, with possible mergers), etc. Dust, thermal conduction and viscosity are believed to be less important.

As a result, the computer codes recently started to produce realistic galaxies. While the details vary (for example, the formation of spirals critically depends on how stellar feedback regulates star formation), different codes presently seem to agree on a number of predictions. The overall conclusion seems to be that the ΛCDM model including baryons can account for galaxy formation and that the basic physics that shapes galaxies has been identified. Such codes do contain, however, freely adjustable parameters, including those that control the numerical approximations, such as the spatial resolution and the modelling of the so-called 'sub-grid physics'. As such, there is the usual danger of possibly (over)fitting data to the incomplete or even incorrect models, where adjusting the phenomenological parameters may mimic the effects of something else.

Numerical simulations also include Dark Energy (DE), at least in the form of an accelerated expansion of the underlying Universe, hence entering in the background Hubble parameter at late times. Alternative models of DE are also simulated. The effect on structure formation can be sizable, but it is not always easy to disentangle it from other degenerate effects.

Finally, we mention that numerical simulations can be used to study the nature of Dark Matter, i.e., to explore the deviations from the assumption of the vanilla cold, collisionless DM. For instance, numerical simulations of warm DM, self-interacting DM and fuzzy DM, with or without baryons, have been performed [24]. We will come back to some of the constraints imposed by these simulations in sections 3.3 and 3.4. In this context, a powerful tool is the ETHOS framework [25]. ETHOS (the Effective THeory Of Structure formation) is a scheme in which the different microphysical DM interactions are translated into general effective parameters that modify the perturbation equations discussed in section 1.3.1. Hence in principle performing one ETHOS numerical simulation, with a chosen set of effective parameters, allows to constrain all the microphysical models captured by such a set.

### 1.3.3 CMB acoustic peaks

In this section we discuss how the cosmic microwave background (CMB) carries the imprint of the presence of DM. The chief observable is the CMB power spectrum (middle left panel in fig. 1.5): it is obtained by performing a spherical Fourier transform of the photon temperature field (the anisotropies in the top left panel of the same figure), and then computing the variance by averaging over the $2\ell + 1$ orientations for each value of the angular momentum, $\ell$. This variance is measured and is compared to the predictions of inflationary big-bang cosmology: such comparison allows to infer information on the contents of the Universe, including, crucially, on the presence of DM.

A bit more precisely, the CMB peaks are due to acoustic oscillations of the baryon/photon fluid, eq. (1.15). The positions of such acoustic peaks depend on the DM density (less DM means later radiation-matter equality), and their amplitude depends on the relative amount of DM with respect to normal matter (only the latter undergoes acoustic oscillations). Global fits find that these and other

cosmological observables can be well reproduced by the Standard Model of cosmology including DM ($\Lambda$CDM with adiabatic Gaussian primordial inflationary perturbations), and allow to determine the values of its cosmological parameters. This gives the most accurate determination of the DM density $\Omega_{\rm DM}$ currently available.

Below, we expand on this discussion and show semi-quantitatively how DM affects the shape of the CMB power spectrum. We over-simplify the physics and focus only on the main DM-related features; a complete discussion can be found in [26].

The CMB power spectrum has the same intuitive meaning as the matter power spectrum $P(k)$, section 1.3.1, but now for photon anisotropies. Roughly, $C_\ell$ is the amount of anisotropy at angular size $\theta \sim \pi/\ell$. An angular scale $\theta$ roughly corresponds to wave-number $k \sim \ell H_0$, where $H_0$ is the present Hubble constant. The CMB power spectrum declines as $\ell$ increases, due to 'Silk damping', i.e., the smoothing of small scale structures due to photon diffusion [27]. However, the 2nd and the 3rd CMB peaks have roughly equal size. This suggests that, were one able to remove the Silk damping, the 3rd peak would have been particularly prominent and higher than the 2nd peak. This is the footprint of DM, as we proceed to illustrate.

Inhomogeneities in the photon fluid at a particular position $\boldsymbol{r}$ in the Universe and at time $t$ are fully described by a function denoted $\Theta(\boldsymbol{r}, t, \hat{\boldsymbol{p}}) \equiv \delta T/T$, or equivalently by its Fourier transform $\Theta(\boldsymbol{k}, t, \hat{\boldsymbol{p}})$. Here $T$ is the photon temperature and $\hat{\boldsymbol{p}}$ is the photon direction. This function obeys the following Boltzmann evolution equation [21][12]

$$\dot{\Theta} - i\frac{k}{a}\mu\Theta + \dot{\Psi} - i\frac{k}{a}\mu\Phi = -\dot{\tau}\left[\Theta_0 - \Theta + \mu v_{\rm bk}\right], \tag{1.19}$$

where $\mu = \hat{\boldsymbol{k}} \cdot \hat{\boldsymbol{p}}$ is the angular variable, $\Phi$ is the Newton potential and $\Psi$ is the curvature perturbation, appearing in the metric

$$ds^2 = -(1 + 2\Phi)dt^2 + a^2 d\boldsymbol{x}^2(1 + 2\Psi). \tag{1.20}$$

Finally, $v_{\rm bk}$ is the velocity perturbation of *b*aryonic matter, further discussed below. The left terms describe how photons move in the gravitational background; the right terms describe how photons electromagnetically interact with baryonic matter: the optical depth $\tau$ is defined later. It is convenient to expand $\Theta$ in multipoles in terms of the Legendre polynomials $\mathcal{P}_\ell$ in the variable $\mu$. The multipoles are denoted as $\Theta_\ell$ and defined as

$$\Theta_\ell = i^\ell \int_{-1}^{1} d\mu \frac{1}{2}\mathcal{P}_\ell(\mu)\Theta(\mu). \tag{1.21}$$

The first two moments, the monopole $\Theta_0(\boldsymbol{k}, t)$ and the dipole $\Theta_1(\boldsymbol{k}, t)$, describe respectively the overall density inhomogeneity and the velocity of the photon fluid, while the higher moments have a less intuitive meaning.

In other words, the evolution of the perturbations in the photon fluid can be described either via the whole function $\Theta(\boldsymbol{k}, t, \hat{\boldsymbol{p}})$ or via an infinite array of multipoles $\Theta_\ell(\boldsymbol{k}, t)$ obeying an infinite array of (coupled) equations derived from eq. (1.19). We now discuss which equations these functions obey

---

[12] Deriving this equation from first principles is well beyond our intended scope. The reader is invited to consult the classic books in [21]. Note that here we also neglect the photon polarization. It is as well worth mentioning that our notation differs from the one used by Dodelson [21]. First, we denote the Newton potential with $\Phi$, consistent with section 1.3.1 (and with Kolb & Turner and Weinberg [21]), and the curvature perturbations with $\Psi$, hence $\Phi \leftrightarrow \Psi$ when comparing our formalism with Dodelson's. Second, $k \to -k$ because of a sign difference in the convention of Fourier transforms (see eq. (1.12)). Third, Dodelson denotes with $\dot{} \equiv d/d\eta$ the derivative with respect to conformal time, while in our notation (again consistently with Kolb & Turner and Weinberg [21]) $\dot{} \equiv d/dt$ and $' \equiv d/d\eta$, hence $X' = a\dot{X}$ in our notation. Finally, in this section we use the $(-+++)$ metric, see eq. (1.20), which is convenient in cosmology and thus eases comparison with the literature. In the later sections we will use the $(+---)$ metric, convenient for particle physics.

during the various phases of the Universe. In particular, a key moment is the epoch of recombination, $a_{\text{recomb}} \approx 1100$, i.e. the point in time when the electrons, protons and neutrons in the plasma formed bound hydrogen and helium atoms. The photons experience a very different evolution before and after this moment. Before recombination the photons constantly scatter on the charged particles, forming a single *tightly coupled baryon-photon* fluid which is non-relativistic. For such a fluid, the Boltzmann equations can be truncated, i.e. only the first two moments introduced above, $\Theta_0(\boldsymbol{k}, t)$ and $\Theta_1(\boldsymbol{k}, t)$, are needed, while higher moments can be neglected. After recombination, photons free-stream from the *surface of last scattering*. They are a relativistic fluid and the higher multipoles have to be reintroduced, as we will do later.

**Tightly coupled regime**

In the tightly coupled regime, the evolution of the perturbations in the photon fluid is governed by the two coupled Boltzmann equations for $\Theta_0$ and $\Theta_1$, which are in turn coupled to the evolution equations for matter. The system reads[13]

$$\begin{cases} \dot{\delta}_k - i\frac{k}{a}v_k + 3\dot{\Psi}_k = 0, \\ a\dot{v}_k + \dot{a}v_k - ik\Phi_k = 0, \end{cases} \qquad \text{DM} \qquad (1.22)$$

$$\begin{cases} \dot{\delta}_{\text{b}k} - i\frac{k}{a}v_{\text{b}k} + 3\dot{\Psi}_k = 0, \\ a\dot{v}_{\text{b}k} + \dot{a}v_{\text{b}k} - ik\Phi_k = a\frac{\dot{\tau}}{R}\big[3i\Theta_1 + v_{\text{b}k}\big], \end{cases} \qquad \text{baryons} \qquad (1.23)$$

$$\begin{cases} \dot{\Theta}_0 - \frac{k}{a}\Theta_1 + \dot{\Psi}_k = 0, \\ \dot{\Theta}_1 + \frac{k}{3a}\Theta_0 + \frac{k}{3a}\Phi_k = \dot{\tau}\left[\Theta_1 - \frac{iv_{\text{b}k}}{3}\right], \end{cases} \qquad \text{photons} \qquad (1.24)$$

with extra equations for neutrinos and for gravity (described by two scalar potentials $\Phi$ and $\Psi$ in the relativistic treatment) which we do not write down for simplicity. Here $R = 3\rho_{\text{b}}/4\rho_\gamma$ and $\tau = \int d\eta\, n_e \sigma_{\text{T}} a$ is the optical depth, expressed in terms of the number density of electrons $n_e$, of the Thomson cross section $\sigma_{\text{T}}$ and of proper time $\eta$ (see appendix C). A few comments are in order. The equations for matter (baryons or DM) should look familiar: they correspond to those derived in eq. (1.13), with a couple of differences: we are here considering the full relativistic treatment, so that the perturbation to curvature $\Psi_k$ now appears (the gravitational potential $\Phi$ is still present); we made the coupling between baryons and photons explicit with the Thomson scattering term (right side of eq. (1.23)). As expected, the only difference between DM and baryons is the coupling to photons. The right side of the second equation in (1.24) is again the Thomson scattering term that couples photons to baryons.

The above equations show that photons are coupled to matter in two ways: via electromagnetism, as expressed by Thomson scattering, and via gravity, as expressed by the gravitational potentials $\Phi$ and $\Psi$ that appear in the equations for all fluids. The coupling to gravity accounts for the physical effect that photons get red-shifted (blue-shifted) when climbing out of (fall in) the gravitational potential wells created by matter. The crucial point is thus that the CMB photons are separately sensitive to *matter that gravitates* and to *matter that also has an electric charge*. In this resides its ability of measuring the densities of DM and baryonic matter separately. Rearranging the above equations shows, qualitatively, how this works in practice.

---

[13]The passages to obtain the equations for $\Theta_0$ and $\Theta_1$ from the eq. (1.19) for $\Theta$ consist in multiplying eq. (1.19) by $\mathcal{P}_\ell$ and doing the integrals over $\mu$ as per the definition of the multipoles in eq. (1.21), using when needed the recursive relation among the Legendre polynomials: $(\ell + 1)\mathcal{P}_{\ell+1}(\mu) = (2\ell + 1)\mu\mathcal{P}_\ell(\mu) - \ell\mathcal{P}_{\ell-1}(\mu)$ [21].

In the tightly coupled limit of large $\dot\tau \to \infty$, the equation for $v_b$ is approximatively solved by $v_{bk} \simeq -3i\Theta_1$ and the equation for the baryonic velocity $v_b$ can be approximately recast as

$$\left[\Theta_1 - \frac{iv_{bk}}{3}\right]\dot\tau \simeq -R\left[\dot\Theta_1 + H\Theta_1 + \frac{k}{3a}\Phi_k\right]. \tag{1.25}$$

This can be used to eliminate matter from the second equation for photons, eq. (1.24), obtaining

$$\dot\Theta_1 + H\frac{R}{1+R}\Theta_1 + \frac{k}{3a(1+R)}\Theta_0 = -\frac{k}{3a}\Phi_k. \tag{1.26}$$

It is convenient to convert time derivatives in the photon equations into derivatives with respect to the conformal time $\eta$, which we denote by $'$:

$$\begin{cases} \Theta_0' - k\Theta_1 = -\Psi_k', \\ \Theta_1' + \frac{a'}{a}\frac{R}{1+R}\Theta_1 + \frac{k}{3(1+R)}\Theta_0 = -\frac{k}{3}\Phi_k. \end{cases} \tag{1.27}$$

These can be combined in a single second order equation for the photon monopole $\Theta_0$, by applying the derivative with respect to $\eta$ to the first equation in (1.27):

$$\Theta_0'' + \frac{a'}{a}\frac{R}{1+R}\Theta_0' + \frac{k^2}{3(1+R)}\Theta_0 = -\frac{k^2}{3}\Phi_k - \frac{a'}{a}\frac{R}{1+R}\Psi_k' - \Psi_k''. \tag{1.28}$$

This is the familiar equation of a damped, forced harmonic oscillator. The damping (the second term on the left) is, as usual, due to the expansion of the Universe, encoded in the prefactor $a'/a = aH$. For simplicity, we will neglect this term in the following. If we could also neglect the forcing (the terms on the right), the equation would read

$$\Theta_0'' + \frac{k^2}{3(1+R)}\Theta_0 = 0, \qquad \text{giving} \qquad \Theta_0 = c_1 \cos\left(\frac{k\eta}{v_s}\right) + c_2 \sin\left(\frac{k\eta}{v_s}\right), \tag{1.29}$$

where $v_s = 1/\sqrt{3(1+R)}$ is the speed of sound of the photon-baryon fluid (if the baryon density is negligible, it gives the standard speed of sound for relativistic fluid, $v_s \to 1/\sqrt{3}$). The solution for $\Theta_0$ is thus an oscillating function of $k\eta$, with all the peaks and troughs having the same amplitude (as is obvious for the sine and cosine functions). Importantly, the oscillation period in $k$ depends on $v_s$, which in turn depends on $\rho_b$ through $R$ (the presence of the non-relativistic baryons makes the fluid heavier and the sound speed lower). The separation between the *acoustic peaks* in the CM is thus going to be influenced by the amount of baryonic matter in the early Universe.

When the forcing provided by gravity is included, $\Theta_0$ no longer oscillates around 0, but rather around the offset provided by the forcing term. Including, for example, just the $\Phi_k$ contribution, the approximate solution becomes

$$\Theta_0 = c_1 \cos\left(\frac{k\eta}{v_s}\right) + c_2 \sin\left(\frac{k\eta}{v_s}\right) - (1+R)\Phi_k. \tag{1.30}$$

Jumping a bit ahead in the argument, we anticipate that we will 'take the square' of functions like these. After doing so, the acoustic peaks no longer have the same height. This is precisely the feature observed in the present day CMB correlations, and thus in the $C_\ell$. That is, the gravitational forcing term (right side of eq. (1.28)), which depends on the amount of DM, controls the relative height between even and odd peaks in $\Theta_0^2(k)$ (and thus in CMB anisotropies), giving a measure of total DM in the Universe.

**Free streaming regime**

In order to proceed with the narration, we need to consider next what happens after recombination, in the regime where photons free stream. For that, we go back to eq. (1.19), which we can rearrange as

$$\Theta' - (ik\mu - \tau')\Theta = \hat{S}, \tag{1.31}$$

with the source function $\hat{S}$ defined as $\hat{S} = -ik\mu\Phi - \tau'\Theta_0$. To get here, we have moved to conformal time, neglected for the moment the time dependence of the gravitational potentials and neglected $v_{bk}$, the dipole of the baryonic matter density. We then find a solution of eq. (1.31) in conformal time and expand it in Legendre polynomials (which causes the spherical Bessel function $j_\ell$ appear). We finally get (see [21]) to the following result, which provides the values of the multipoles of the photon field at the present day (i.e. at $\eta_0$):

$$\Theta_\ell(k, \eta_0) = \int_0^{\eta_0} d\eta \; g(\eta) \big[\Theta_0(k, \eta) + \Phi(k, \eta)\big] \; j_\ell[k(\eta_0 - \eta)]. \tag{1.32}$$

Here the visibility function $g(\eta) = -\tau' \exp[-\tau(\eta)]$ gives the probability for a photon to last scatter at conformal time $\eta$. It is peaked at $\eta \approx \eta_{\mathrm{recomb}}$: for $\eta \ll \eta_{\mathrm{recomb}}$ it is very unlikely that the photon would not continue to scatter in the tightly coupled photon-baryon fluid, while for $\eta \gg \eta_{\mathrm{recomb}}$ the Universe is transparent. Thus in both limits $g(\eta) \to 0$ and, to a good approximation,

$$\Theta_\ell(k, \eta_0) \approx \big[\Theta_0(k, \eta_{\mathrm{recomb}}) + \Phi(k, \eta_{\mathrm{recomb}})\big] j_\ell[k(\eta_0 - \eta_{\mathrm{recomb}})]. \tag{1.33}$$

This solution has an intuitive meaning. The photon perturbations do not grow after recombination — the gravitational potentials in the Universe are too weak to trap photons and these basically free stream, giving us a snapshot of the Universe at the surface of last scattering. The CMB temperature field is controlled by the sum of kinetic and potential energy, $\Theta_0(k, \eta_{\mathrm{recomb}}) + \Phi(k, \eta_{\mathrm{recomb}}) = -\delta(k, \eta_{\mathrm{recomb}})/6$, i.e., the temperature of the photons at present day includes the fact that they needed to climb out of the potentials they were in at the time of recombination. Furthermore, the perturbations in this total energy of photons is controlled by the DM perturbations $\delta$ (with a negative sign, as hotter regions now correspond to under-dense regions in DM fluid at the time of recombination). The spherical Bessel function $j_\ell[k(\eta_0 - \eta_{\mathrm{recomb}})]$ just encodes how much anisotropy there is on angular scale $1/\ell$ from a pure plane wave with wavenumber $k$. For large $\ell$, $j_\ell(x) \to (x/\ell)^{\ell-1/2}/\ell$ and thus this gives relevant contributions to CMB anisotropies only for $\ell \sim k\eta_0$ ($\eta_{\mathrm{recomb}}$ can be neglected since $\eta_0 \gg \eta_{\mathrm{recomb}}$).

One last formal step is needed to make contact with the $C_\ell$ featured in the CMB power spectrum and mentioned at the beginning. One can show that explicitly

$$C_\ell = \frac{2}{\pi} \int_0^\infty dk \, k^2 P(k) \left| \frac{\Theta_\ell(k)}{\delta(k)} \right|^2, \tag{1.34}$$

where $P(k)$ is the matter power spectrum and $\delta$ the matter perturbations. This substantiates the qualitative statements that the $C_\ell$'s are the 'averages of the squares' of the $\Theta_\ell$'s, provided by eq. (1.33) in terms of the $\Theta_0$ at the time of recombination, in turn determined by the considerations in the tight coupling regime (eq. (1.30)).

Eq. (1.33) is approximate, and some corrections are important. The missing dipole contribution is smaller and out of phase with the monopole. Including it, gives higher throughs in $|\Theta_\ell|^2$ (which give the $C_\ell$ as just seen) and thus the peaks become less pronounced. The time dependence of the gravitational potentials enhances the $C_\ell$ at low $\ell$, up to around the first peak at $\ell \approx 200$ (the so called *integrated Sachs-Wolfe* effect). It reflects the fact that the energy density stored in radiation is not entirely negligible at the time of recombination. Finally, the photon diffusion results in damping of all high $k$ (high $\ell$) modes.

A photon undergoes a random walk when it scatters on a sea of electrons, and therefore diffuses by a comoving distance $\lambda_D \sim 1/a\sqrt{n_e \sigma_T H}$ during a Hubble time $H$. Over distances smaller than $\sim \lambda_D$ the photons restore the temperature of the baryon-photon fluid to a single mean temperature, erasing any anisotropies. This effect roughly amounts to a $\sim \exp(-\lambda_D^2 k^2)$ suppression of the solution in eq. (1.33), visibly suppressing anisotropies for $k\eta_0 \gtrsim 500$.

The above simplified analysis cannot capture all the complexity of the real Universe, but the main message is clear: the relative heights of two neighbouring peaks in the CMB power spectrum (in particular the 2$^{\text{nd}}$ and the 3$^{\text{rd}}$ peaks) carry information about the abundance of the kind of matter that was coupled by electromagnetism to photons (baryons, which provide the term with $R$) and the abundance of matter that provides the gravitational forcing $\Phi$. The latter turns out to be much larger than the former, providing the conclusive evidence for the existence of DM. If the Universe contained baryonic matter only, the 3$^{\text{rd}}$ peak would be suppressed, as shown in the bottom left plot of fig. 1.5, in disagreement with the data.

### 1.3.4   Cosmological bounds on Dark Matter properties

As we saw in sections 1.3.1 and 1.3.3, the large scale structure (LSS) and CMB data allow to test DM behaviour. Global fits prefer the simplest model of *cold non-interacting dark matter with Gaussian adiabatic perturbations*. Below, we summarize what this means and present the current bounds.

One can modify the Boltzmann equations for cosmological perturbations in sections 1.3.1 and 1.3.3, replacing DM with a more complicated dark fluid with nonzero sound speed $v_s$ (making DM not cold) and/or viscosity (making DM interacting) and/or a modified equation of state $w = \wp/\rho \neq 0$ (making DM no longer behave as nonrelativistic matter). Global fits to CMB data give bounds at the few $10^{-3}$ level on these parameters [28].

Concerning adiabaticity — Boltzmann equations in section 1.3.3 have to be solved for each component and for any scale $k$ starting from initial conditions at the time much before the matter/radiation equality. CMB data are well fitted assuming *adiabatic* initial conditions: one primordial inhomogeneity common to all components

$$\delta_k = \delta_{\text{b}k} = 3\Theta_0(k) = 3N_0(k), \ldots \tag{1.35}$$

The factor 3 for photons and neutrinos arises just because $\Theta_0$ and $N_0$ are defined as inhomogeneities in the temperature rather than in the density. Eq. (1.35) means that the density of each component is proportional to the total density, so that the overall effect can be described by a geometric primordial curvature (or, equivalently, by the gravitational potential). Such adiabatic initial conditions arise in minimal inflation models, where all primordial inhomogeneities are generated by quantum fluctuation of one 'inflaton' scalar field. As usual, quantum mechanics provides predictions for the statistical distributions of these fluctuations, rather than specific outcomes of individual measurements. That is why all the observables are averaged over space, or, equivalently, over the Fourier modes with different orientations.

Concerning Gaussianity — in sections 1.3.1 and 1.3.3 we focused on the simplest two-point correlation functions, $\langle a_{\ell m} a_{\ell' m'} \rangle = \delta_{\ell \ell'} \delta_{mm'} C_\ell$ in section 1.3.3 and $\langle \delta_k \delta_{k'} \rangle = (2\pi)^3 P(k)\,\delta^3(\boldsymbol{k} - \boldsymbol{k}')$ in section 1.3.1. The reason is that the quantum fluctuations of a free scalar field with negligible interactions (such as the inflaton with its quasi-flat potential) follow a Gaussian distribution (since this is the ground state of a free harmonic oscillator). Gaussian primordial inhomogeneities, on the other hand, are fully determined by the two-point correlation functions, compactly described by a single power spectrum. Gaussian inhomogeneities fit the CMB and LSS data. However, this is not the only possibility. The inflaton could have interactions, resulting in **non-Gaussianities** in the probability distribution for the

curvature perturbation $\zeta$,

$$\wp(\zeta) \sim \exp\left[-\frac{\zeta^2}{2\langle\zeta^2\rangle} - \frac{\langle\zeta^3\rangle\zeta^3}{\langle\zeta^2\rangle^3} + \cdots\right] = \exp\left[-\frac{\zeta^2}{2\langle\zeta^2\rangle}\left(1 + f_{NL}\zeta + \cdots\right)\right] \tag{1.36}$$

where the variety of non-vanishing 3-point functions is loosely characterised by a $f_{NL} \sim \langle\zeta^3\rangle/\langle\zeta^2\rangle^2$ parameter. Single-field inflation can reproduce data assuming a nearly-flat inflaton potential, corresponding to a small slow-roll parameter $\epsilon \sim 0.01$, so that the nearly non-interacting inflaton produces $P_\zeta \sim \langle\zeta^2\rangle \sim H_{infl}^2/M_{Pl}^2\epsilon$ and a small $f_{NL} \sim \epsilon$ [29]. This is much below current bounds on non-Gaussianities implied by the CMB data, $f_{NL} \lesssim 10^{1-2}$ [3]. Future CMB and large scale structure data can improve the sensitivity to $f_{NL}$ by one and two orders of magnitude, respectively. Even lower values can maybe be probed by futuristic 21 cm observations. Still, it remains difficult to devise non-minimal inflationary models that provide signals in $f_{NL}$.

Returning to the issue of adiabaticity — in the inflationary context the extra inhomogeneities can arise, for example, from independent quantum fluctuations of some other fields, which happen to be light enough during the inflationary period. From a phenomenological point of view it is convenient to consider the so called **iso-curvature** fluctuations. These violate eq. (1.35), by ascribing different inhomogeneities to different species, but without affecting the inhomogeneity in the total energy density, and thereby in the curvature. We are interested in isocurvature fluctuations in the primordial DM density $\Delta_{iso}(k) = \delta\rho_{kDM}/\rho_{DM}$, corresponding to the power spectrum $\Delta_{iso}^2 = k^3 P_{iso}/2\pi^2$. The fluctuations in DM density on scales $1/k$ would then differ from those in the SM particles. Global fits to Planck data demand that adiabatic fluctuations are dominant:

$$\frac{P_{iso}(k)}{P_{iso}(k) + P_{adi}(k)} < 0.038 \qquad \text{at 95\% C.L. at } k = 0.05/\text{Mpc [3]}. \tag{1.37}$$

The bound applies to fluctuations on scales $1/k$ that entered the horizon around matter/radiation equality, such that $\Delta_{adi}^2 \approx 2.2 \, 10^{-9}$ is well probed by the CMB. Iso-curvature fluctuations on smaller, subcosmological, scales are unconstrained (e.g., on galactic scales). Loosely speaking, the fact that DM has the same fluctuations as normal matter suggests some early connection between the two, and restricts the viable DM production mechanisms, see the discussion in chapter 3.

So far we assumed that DM is non-interacting. If some extra interaction, other than gravity, acts on DM particles, then Eq. (1.22), which describes the gravitational clustering of DM, gets modified.

○ If **DM interacts with neutrinos** [30][14] this generically suppresses DM primordial density fluctuations through collisional damping, and thereby leaves noticeable signatures in the CMB anisotropy power spectra and in the matter power spectrum. A rough summary is that current data constrain the DM/$\nu$ interaction cross section to be $\sigma_{DM/\nu} \lesssim 10^{-32} (M/\text{GeV}) \, \text{cm}^2$ (assuming absence of temperature dependence), with different studies finding an order of magnitude weaker or stronger bounds.

○ Similarly, the possibility that **DM interacts with photons** [31] remains viable as long as the interaction is sufficiently weak. The phenomenological consequences are similar to the case of interactions with neutrinos (to first approximation neutrinos and photons are simply two forms of radiation at the time of CMB). A rough summary is again that current data constrain the DM/$\gamma$ interaction cross section to be $\sigma_{DM/\gamma} \lesssim$ few $10^{-33} (M/\text{GeV}) \, \text{cm}^2$ (assuming there is no temperature dependence). This is not particularly constraining: in the case where interactions with photons are due to DM carrying a small electric charge, significantly stronger constraints do exist, and are discussed in section 3.3.2.

---

[14]This is not to be confused with DM/neutrino interactions at higher energies in current astrophysical systems, discussed in section 6.3.1.

○ Another possibility is an **extra long-range force** acting on DM (either only on DM or possibly also on other fields). Such a force could arise, for example, if DM couples to an extra scalar with ultra-light (sub-Hubble) mass. Global fits constrain the strength of such speculative dark force to be more than 100 times weaker than gravity [32].

## 1.3.5   Big Bang Nucleosynthesis

The determination of the density of normal matter obtained from the global fits to the CMB data ($\Omega_{\rm b}h^2$ on page 15) agrees with the independent precision determination derived from the Big Bang Nucleosynthesis (BBN). BBN is a theoretical framework which describes the synthesis of light elements in the Universe (the bulk of D and $^4$He and a good portion of $^3$He and $^7$Li), starting from their building blocks: the free protons and neutrons present in the primordial plasma. More precisely, BBN is sensitive to the baryon-to-photon ratio $\eta \equiv n_{\rm b}/n_\gamma$, because photons break nuclei, delaying their formation temperature. The measurements give [5]

$$\eta = (6.2 \pm 0.4)\ 10^{-10} \qquad \Rightarrow \qquad \Omega_{\rm b}h^2 = 0.022 \pm 0.002, \tag{1.38}$$

in agreement with the CMB value ($\Omega_{\rm b}h^2$ on p. 15). To derive the second relation above, we used $\Omega_{\rm b} \equiv \rho_{\rm b}/\rho_{\rm crit} = \eta\, n_\gamma^0 m_p/\rho_{\rm crit}$, where $m_p$ is the proton mass and $n_\gamma^0$ the present photon number density.

The concordance between the two determinations of the baryon density is seen as a major success of the standard model of cosmology, giving confidence that the evolution of the Universe is well understood within this model. BBN probes an earlier phase of the Universe than the CMB does, i.e., for significantly larger temperatures $T \sim$ MeV, compared to $T \sim$ eV for CMB. The fact that $\Omega_{\rm b}$ in (1.38) satisfies $\Omega_{\rm b} < \Omega_{\rm matter} \simeq 0.30$, where $\Omega_{\rm matter}$ includes all forms of matter, provides further evidence for DM. This is required to behave as a non-baryonic matter, and is needed for the structure formation as discussed in section 1.3.1. The result from BBN in (1.38) has been especially important historically, before the CMB and LSS measurements became precise enough to be able to pin down by themselves the relative proportions of baryons and DM.

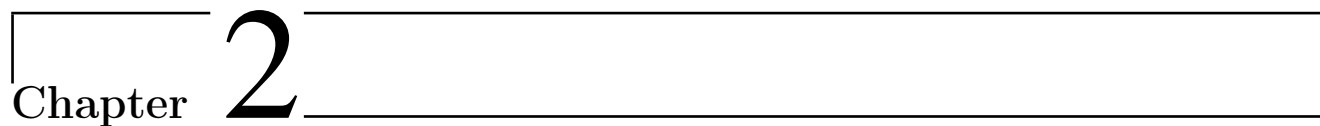

# Chapter 2

# Where? Dark Matter distribution in the Galaxy and the Universe

As discussed in chapter 1, the average DM density in the Universe is nonzero, see eq. (1.1). However, there is much more to DM than just this average value. Because of structure formation, the distribution of DM in specific systems differs significantly from the average cosmological value. In particular, in order to interpret the results of direct and indirect DM detection searches (discussed in chapters 5 and 6) one needs the DM density distribution, $\rho(\boldsymbol{x})$, and the DM speed distribution, $f(\boldsymbol{x}, \boldsymbol{v})$, in the Milky Way, as well as in other galaxies and in a variety of other astrophysical systems. Presently, only educated guesses are available for these quantities, with significant uncertainties. In this chapter we discuss the methods with which such guesses are derived, and their results. We start with two general statements, which are simple, yet at the same time also quite powerful:

○ DM tends to be **roughly spherically distributed** in gravitationally bound systems. The initial conditions in structure formation generally predict nearly spherical distributions. Normal baryonic matter then collapses in the gravitational well, during which it dissipates energy and cools down, resulting in the spinning disks exhibited by many galaxies (see section 2.4). Unless DM has similarly large scattering cross sections, it has negligible interactions and thereby negligible dissipation. Neglecting the gravitational effects of visible matter, DM therefore conserves a spherical distribution, so that one can reasonably assume that the DM density $\rho(r)$ in astrophysical systems is, to a good approximation, a function of only the radial coordinate $r$.

○ DM is **non-relativistic** in most systems of interest.[1] For instance, DM particles bound to our galaxy must have a velocity below the escape velocity, $v_{\text{esc}} \approx 500\,\text{km/s}$. Similar considerations hold for other systems such as dwarf galaxies, galaxy clusters, etc, with the appropriate $v_{\text{esc}}$.

Going beyond the above general statements, $\rho(r)$ and $f(v)$ for any given system could in principle be determined in three conceptually independent ways: A) **observationally**, by precisely tracing stellar kinematics in a galaxy (or galactic kinematics in a cluster), inferring the underlying gravitational potential, and thereby the density and velocity distribution of matter responsible for the observed motion of tracers; B) via **theoretical methods**, describing (semi-)analytically the process of gravitational collapse and its end result; C) with **$N$-body numerical simulations**.

In practice, A) is too imprecise to provide the full $\rho(r)$, but it helps in determining the local DM density; B) faces formal and technical difficulties, due to the non-linear nature of the problem and due

---

[1] A notable exception are DM particles in a close proximity of a dense massive body such as a neutron star. In such cases DM particles acquire relativistic speeds and can even transfer their large kinetic energy to the star, see, e.g., section 6.10.4.

to the presence of baryonic physics, yet it provides valuable insights into the complex processes involved. Therefore, the current most popular approach relies mostly on C), with an admixture of other strategies. Usually, this is a two step procedure:

1. Guess the functional forms of $\rho(r)$ and $f(v)$ in terms of a minimal number of free parameters, on the basis of (most often) simulations, or theory, or observations.

2. Determine the free parameters in terms of solid observations of DM in our Galaxy, or in other galaxies.

We will present the results of this procedure in section 2.2 and 2.3 for the density and speed distributions respectively. Finally, in section 2.4 we discuss how things can be further refined, going beyond the 'dark spherical cow' limit. Before doing that, however, it is worth elaborating on the three methods mentioned above.

# 2.1    Methods to determine the DM distributions

## 2.1.1    Observations

As discussed in section 1.1.1, the basic observation that the galactic rotation curves flatten, allows to infer $\rho(r) \propto 1/r^2$ at larger radii $r$, where DM dominates. One can use the same idea to try to extract $\rho(r)$ also at smaller $r$: objects in circular orbits around the galactic center are essentially tracers of the total gravitational potential, hence of the distribution of matter. The mismatch between the rotation curve obtained with such tracers and the one expected from the visible matter in the galaxy[2] must be accounted for by Dark Matter. The DM density distribution can thus be obtained by fitting an appropriately parametrized function to the total rotation curve [33]. This is often referred to as the *global method* to determine the DM distribution (as opposed to the *local* method, discussed below, which is less ambitious, and only aims to determine the DM density at the location of the solar system, and about 1 kpc around it).

The method is conceptually simple. To obtain the predictions for the rotation curves, the Poisson equation for the Newton potential $\phi$, $\nabla^2 \phi = 4\pi G \rho$, is first solved assuming as a source only the observed mass distributions due to baryons in the stars and from those in the inter-stellar gas, $\rho = \rho_b = \rho_{\text{star}} + \rho_{\text{gas}}$, taking into account their approximate cylindrical (rather than spherical) geometry $r, z, \theta$. The Newton's equation for a test mass in a gravitational field, $v^2/r = -\partial\phi/\partial r$, then predicts the rotation velocity $v(r) = (v_{\text{star}}^2 + v_{\text{gas}}^2)^{1/2}$ in the galactic plane, at $z = 0$, where these are measured.[3] Comparison with data for $v(r)$ shows that an extra 'dark' contribution is needed in order to reproduce the observations,

$$v^2 = v_{\text{star}}^2 + v_{\text{gas}}^2 + v_{\text{dark}}^2, \tag{2.1}$$

which then allows to determine the DM contribution by subtracting the predictions from observations.

However, a number of difficulties arise. First, stellar kinematical data still carry significant uncertainties, both intrinsic to the measurements and due to their interpretation (also in the Milky Way, related, e.g., to the knowledge of the precise distance of the Sun from the galactic center and of the Sun's circular velocity). Second, the results of the mismatch fit, mentioned above, depends crucially on what is being assumed for the visible component of the galaxy. More precisely, one has to design a *model of the galaxy* as a system composed of a bulge, a possible bar, a disk, a gas halo, and possibly even a smaller disk at

---

[2]Here, 'galaxy' refers to both the Milky Way, as well as to any other galaxy, including the satellites of the Milky Way, for which one is trying to determine the DM distribution.

[3]The velocities orthogonal to the galactic plane and the radial velocities provide extra information, in principle. In practice, however, they are not measured well enough.

the core of the bulge..., each of these appropriately modeled using several observation-based configurations. The gas density is extracted, e.g., from the 21 cm maps, while the stellar models are needed to convert observed luminosity into mass density. The uncertainties on these *baryonic morphologies* are still sizeable, despite the recent improvements due to the fact that data mostly based on optical observations are now being replaced by the near-IR data, which need only minor modeling corrections. Finally, since the baryonic matter dominates the gravitational potential in the inner few kpc of galaxies (the inner $\sim 5$ kpc for the Milky Way), the contribution of DM is intrinsically very difficult to determine in that region.

In practice, therefore, observational methods are sufficient to affirm the presence of DM in galaxies and to get a rough idea of its large scale distribution, but are at present not sufficient for more precise determinations.

## 2.1.2 Gravitational collapse of collision-less DM

The formation of a Dark Matter galactic halo is a conceptually well defined theoretical problem: start from a roughly spherical slight over-density made of collision-less and dissipation-less bodies, and let it evolve under gravity. The system starts collapsing under its own weight (and at some point the inhomogeneity grows beyond the computable linear approximation) but *the collapse self-limits without reaching a point with infinite density*. This happens because DM is collision-less and cannot dissipate energy, so during the collapse its gravitational potential energy has to be converted into kinetic energy of the DM particles involved. The sphere therefore eventually relaxes to a self-gravitating quasi-static structure supported by random motions of DM particles. The whole process is referred to as *virialization* and the resulting system is *virialized*, i.e., its energy is distributed between the potential energy, $V$, and kinetic energy, $K$, according to the virial theorem $\langle K \rangle = -\frac{1}{2}\langle V \rangle$ (already mentioned in section 1.2) [34].

The virial theorem provides enough information to estimate the typical scales involved. For simplicity, we consider a spherical DM over-density with homogeneous density of total mass $\mathcal{M}_{\rm tot}$ (ellipsoidal collapse models with non-constant density are also considered in the literature). The main phases, discussed below, are illustrated in fig. 2.1. We define $\rho_{\rm ta}$ as the homogeneous density at the instant of *turnaround*, i.e., at the moment at which the over-density stops following the expansion caused by the Big Bang, stalls, and turns around to start collapsing into a gravitationally bound object. This marks the onset of structure formation. Since DM is stalling, its kinetic energy $K_{\rm ta}$ vanishes, and thus its total energy $E_{\rm ta} = K_{\rm ta} + V_{\rm ta}$ is entirely due to its gravitational potential energy

$$V_{\rm ta} = -G \int_0^{r_{\rm ta}} \left( \frac{4\pi r^3}{3} \rho_{\rm in} \right) \frac{4\pi r^2 \, dr \, \rho_{\rm ta}}{r} = -\frac{3}{5} \frac{G\mathcal{M}_{\rm tot}^2}{r_{\rm ta}}, \tag{2.2}$$

where $r_{\rm ta}$ is the size of the sphere at turnaround, and $\rho_{\rm in}$ is the density inside the sphere (in our approximation taken equal to $\rho_{\rm ta}$).

If spherical symmetry were absolutely exact, i.e., if all DM particles were following a perfectly radial trajectory, the collapse would have proceeded toward infinite density, forming a black hole. In reality, the system keeps sphericity only on average and reaches a finite density thanks to the chaotic motion of DM particles, interacting only via gravity. After this *virialization* process the energy is distributed between the kinetic and potential energy, such that $E_{\rm vir} = K_{\rm vir} + V_{\rm vir} = V_{\rm vir}/2$, where the last equality follows from the virial theorem. Since virialization conserves the total energy, the potential energy $V_{\rm vir} = -3G\mathcal{M}_{\rm tot}^2/5r_{\rm vir}$ (we still approximate the density as constant, $\rho_{\rm vir} = \mathcal{M}_{\rm tot}/(\frac{4}{3}\pi r_{\rm vir}^3)$, where now $r_{\rm vir}$ is the radius of the virialized halo) therefore implies $r_{\rm vir} = r_{\rm ta}/2$. That is, the collapse results in a relaxed halo half the size of the turn-around bubble, with an order of magnitude higher density, $\rho_{\rm vir} = 8\rho_{\rm ta}$.[4] The mean square speed

---

[4] This means that the collapse of collision-less DM does *not* form black holes. The situation is strikingly different for normal matter. Since normal matter has large scattering cross sections, it is opaque to photons, so that these are emitted into surroundings only by surfaces of compact objects and not by the whole volumes. A

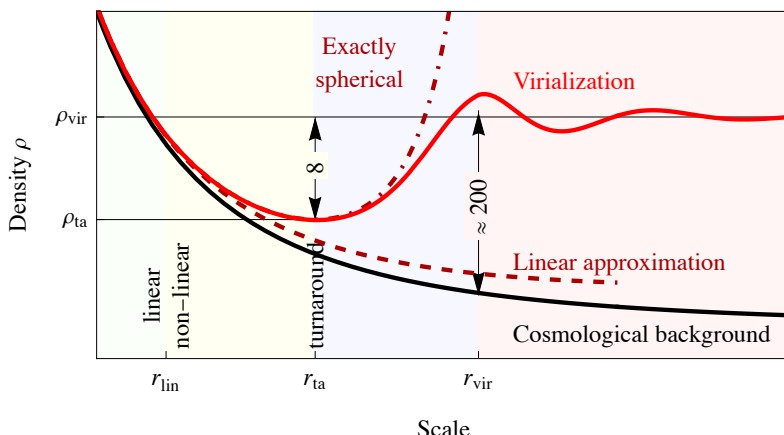

Figure 2.1: *Qualitative illustration of the main phases of a nearly-spherical **gravitational collapse**. The dot-dashed curve shows the result assuming an unrealistic exact spherical symmetry; the continuous red curve shows the realistic result; the dashed curve shows the result within linear approximation, which remains close to the cosmological background (bottom black curve).*

of virialized particles is $\langle v^2 \rangle = 2K_{\mathrm{vir}}/\mathcal{M}_{\mathrm{tot}} = |V_{\mathrm{vir}}|/\mathcal{M}_{\mathrm{tot}} = 3G\mathcal{M}_{\mathrm{tot}}/5r_{\mathrm{vir}}$. Solving Newton equations for a homogeneous density in an expanding matter-dominated universe shows that the average density of the formed halo $\rho_{\mathrm{vir}}$ is roughly 200 times the surrounding cosmological density

$$\bar{\rho} = \frac{3H^2}{8\pi G} = \frac{1}{6\pi\, G\, t_{\mathrm{vir}}^2}, \tag{2.3}$$

where $t_{\mathrm{vir}}$ is the time marking the end of virialization [34] (here we used eq. (C.7) and eq. (C.14) for matter domination). Quite often the *virial radius* is thus defined as the radius within which the average density of the halo is 200 times larger than the cosmological density, and is denoted as $r_{200}$. The mass inside $r_{200}$ is taken as a measure of the total mass of the halo. The reality, at least as seen in numerical simulations, is more involved, but orders of magnitude are correct.

Besides its global properties, one can gain a slightly more refined understanding of the final DM halo (in particular its density and speed distributions, in which we are interested here) by following analytically the collapse and paying particular attention to the *relaxation* mechanisms. There are several such dynamical processes at play, driving the system toward an equilibrium configuration [34]. The important mechanism for collision-less DM particles is the so called *violent relaxation* [35]: the energy of an individual DM particle changes with time due to the fact that the gravitational potential in which these are immersed (generated by the ensemble of DM particles themselves) also changes with time, because of the ongoing collapse. This mechanism works in an efficient way on the short timescale of the collapse, hence the name 'violent'. The total energy of a given DM particle divided by its mass, $\varepsilon = v^2/2 + \varphi$, varies as $d\varepsilon/dt = \partial\varphi/\partial t$, if the gravitational potential is time dependent. The last relation is straightforward to verify,

$$\frac{d\varepsilon}{dt} = \frac{\partial\varepsilon}{\partial\boldsymbol{v}} \cdot \frac{d\boldsymbol{v}}{dt} + \frac{\partial\varepsilon}{\partial\varphi}\frac{d\varphi}{dt} = \boldsymbol{v}\cdot(-\boldsymbol{\nabla}\varphi) + 1\left(\frac{\partial\varphi}{\partial t} + \boldsymbol{v}\cdot\boldsymbol{\nabla}\varphi\right) = \frac{\partial\varphi}{\partial t},$$

---

non-spherical collapse of a large over-density of normal matter tends to fragment, thereby forming many stars, rather than one big object. Stars can then evolve into stellar-mass black holes. How the super-massive black holes form is an area of active research, with one possibility that they form from accretion of dust, gas, as well as stars and stellar-mass black holes.

where the equalities follow trivially from the chain rule, the definition of $\varepsilon$ and from $d\boldsymbol{v}/dt = -\boldsymbol{\nabla}\varphi$, which is the Newton's law, $\boldsymbol{a} = \boldsymbol{F}/m$, for the case of just gravitational interactions.

Whether the particle gains or looses energy is a complex question. One can qualitatively figure out what is going on by thinking of a DM particle that starts collapsing from the outskirts of the halo (which itself will first collapse and then expand, until the particles virialize). During the infall, the DM particle will dive into a deep potential due to the collapsing halo and gain kinetic energy. After crossing the center (unscathed, because of its collision-less nature), it will climb out of a shallower potential, because its fellow DM particles are dispersing, and will therefore spend less than the previously acquired kinetic energy, resulting in a net energy gain. The opposite happens for particles that start from inner locations. For instance, a particle at rest in the center does not move, while as the protohalo collapses, the potential at its center becomes deeper. In general, the gain or loss of energy of a single particle depends on its initial position and initial speed, as well as on the distribution of all other DM, in a complex way [34,35].

The whole process effectively amounts to DM particles undergoing many gravitational interactions. As a consequence, their final velocities are given by a sum of many random contributions. Due to the Central Limit Theorem their energy distribution will tend toward a Gaussian:

$$f(\varepsilon) = \frac{n_0}{(2\pi\sigma^2)^{3/2}} \, e^{-\varepsilon/\sigma^2}, \tag{2.4}$$

where the peculiar form of the normalization factor multiplying the exponential will be clear soon. The exponent is $\varepsilon = \frac{1}{2}\boldsymbol{v}^2 + \varphi$, from which it immediately follows that the energy distribution $f(\epsilon)$ can be interpreted as the distribution of DM speeds, $f(v)$, taking the Maxwell-Boltzmann form

$$f(v) \propto e^{-v^2/2\sigma^2}. \tag{2.5}$$

The uniform 'temperature' (related to velocity dispersion $\sigma$) is for a fully virialized halo given by its total mass.

To show that, we move next to the discussion of the DM density distribution, $\rho(r)$. For spherically symmetric systems there is a unique correspondence between $f(v)$ and $\rho(r)$ (this general relation is known as the *Eddington's formula*, and the art of deriving $\rho(r)$ from $f(v)$ as the *Eddington inversion*). It can be derived by first integrating $f(\varepsilon)$ over all velocities,[5] which then gives an implicit expression for the DM density distribution

$$\rho = M \int_0^\infty dv\, 4\pi v^2 f(\epsilon) = M \int_0^\infty dv\, 4\pi v^2 \, \frac{n_0}{(2\pi\sigma^2)^{3/2}} \, e^{-(v^2/2+\varphi)/\sigma^2} = \rho_0 \, e^{-\varphi/\sigma^2}, \tag{2.6}$$

with $\rho_0 = Mn_0$. Eq. (2.6) gives the gravitational potential in terms of $\rho$. Inserting this expression into Poisson's equation, $\nabla^2\varphi = 4\pi G\,\rho$, written in spherical coordinates, leads to

$$\frac{d}{dr}\left(r^2\frac{d}{dr}\ln\rho\right) = -\frac{4\pi\,G}{\sigma^2}r^2\rho. \tag{2.7}$$

The solution is

$$\rho(r) = \frac{\sigma^2}{2\pi G}\frac{1}{r^2}. \tag{2.8}$$

---

[5]More precisely, what we are actually dealing with is the distribution function $f(\boldsymbol{x},\boldsymbol{v},t)$, defined such that $f(\boldsymbol{x},\boldsymbol{v},t)\,d^3\boldsymbol{x}\,d^3\boldsymbol{v}$ is the probability at time $t$ of finding a particle in the $d^3\boldsymbol{x}\,d^3\boldsymbol{v}$ phase space volume centered at $\boldsymbol{x},\boldsymbol{v}$. For a spherical steady-state system it can be shown that $f$ depends on $\boldsymbol{x},\boldsymbol{v}$ only through the total energy, which is an integral of motion ($f$ is said to be *ergodic*). That is why by integrating $f(\varepsilon)$ over all velocities one obtains the spatial distribution of DM particles, i.e., their number density (the use of $n_0$ in the normalization factor in (2.4) was already the first hint for this). Multiplying it by the DM mass $M$ one then obtains the mass density $\rho$. For more details see, e.g., chapter 4 in Binney & Tremaine [34].

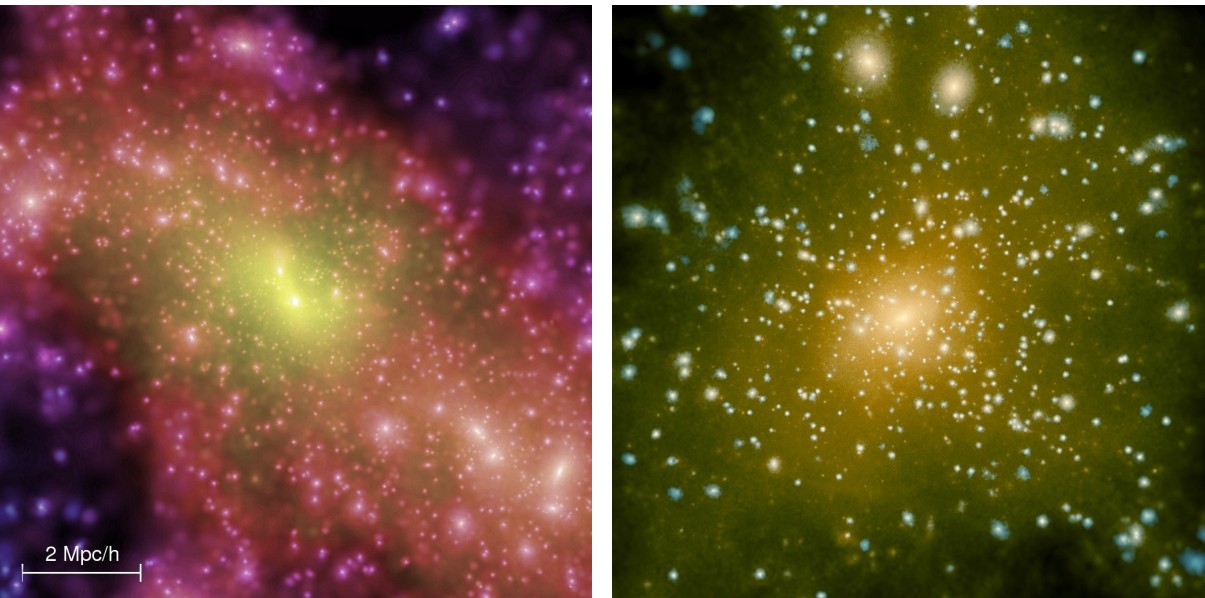

Figure 2.2: **Sub-halos**. *N-body simulations find that DM forms roughly spherical halos, which are then surrounded as well as inhabited by many smaller sub-halos. From simulation realized at the MPI, credit: Max-Planck-Institute for Astrophysics.*

This distribution is the *singular isothermal sphere*. The name comes from the fact that this is the same equilibrium configuration, characterized by a single $r$ independent temperature $T$, to which an ideal gas is driven by collisions and interactions. The role of 'collisions and interactions' is played here by the gravitational scatterings between DM particles and the whole halo, with the effective cross-section given by

$$\sigma_{\mathrm{grav}} = \frac{G\,M\mathcal{M}_{\mathrm{tot}}}{r^2 v^4}. \tag{2.9}$$

In our case, the quantity that is constant with radius is the velocity dispersion $\sigma$: the correspondence with the case of an ideal gas is achieved if one identifies $\sigma^2 = k_B T/M$, where $k_B$ is the Boltzmann constant.

In summary, the classical theory of collapse of isolated, spherical, collision-less and dissipation-less systems predicts equilibrium DM halos that are singular isothermal spheres with exactly Maxwell-Boltzmann speed distributions. This is often called the *Standard Halo Model* (SHM).

In practice, the singular isothermal model is just an idealized approximation. Its density diverges as $r \to 0$, but this is not a dramatic problem since the mass enclosed within radius $r$ is finite, $\mathcal{M}(r) = \int_0^r dr\, 4\pi r^2\, \rho = 2\sigma^2 r/G$. The resulting rotation curve is exactly flat, $v_{\mathrm{circ}}^2(r) = G\mathcal{M}(r)/r = 2\sigma^2$. On the other hand, integrating towards large $r$, the total mass diverges, which is absurd from the astrophysical point of view: the profile must therefore be amended and truncated at some large radius. More generally, many effects can cause deviations from the simple picture that led to isothermal distribution: i) the collapse process can be incomplete and not reach a full equilibrium state, ii) the assumption of spherical symmetry is easily violated by non-radial deformations in the initial conditions, iii) the hypothesis that the system is isolated is not valid for halos embedded in the DM cosmic web: inflows, outflows and mergers cannot typically be neglected, etc.

The Standard Halo Model can be extended to correct for some of these shortcomings, making it more realistic [34, 36], although often at the price of a more involved analytical treatment. Given these complexities and the limitations of purely analytical approaches, numerical simulations have increasingly become an important tool to determine the properties of the DM halos. We discuss these next.

### 2.1.3   Results from $N$-body simulations

Since statistical mechanics does not provide sharp general answers, the gravitational collapse of dark matter is also studied numerically, using the $N$-body simulations [22], which we already introduced in section 1.3.2. While such computer codes do not simulate a specific galaxy (we are especially interested in the Milky Way, or local dwarfs), but rather an ensemble of more or less typical galaxies, their results do show that the DM halos density profiles tend to possess some universal features, as summarized below.

DM halos are produced with sizes that span all simulated scales: from cosmological scales around 100 Mpc down to sub-galactic scales of tens of pc, ranging about 20 orders of magnitude in the enclosed mass $\mathcal{M}$. The number of halos, $n$, as a function of their mass scales roughly as $dn/d\mathcal{M} \propto 1/\mathcal{M}^2$, and will be discussed analytically in section 2.2.5.

To a first approximation, luminous matter follows the distribution of the large scale structure, i.e., galaxies form where matter is dense. However, galaxies do not trace *exactly* the underlying DM distribution, because galaxy formation proceeds with an efficiency that depends on the halo mass. The relationship between the mean over-density of galaxies $\delta_{\text{gal}}$ and the mean overdensity of the total mass $\delta_{\text{tot}}$ (dominated by DM) is complicated. Their ratio, the so called *bias* $b = \delta_{\text{gal}}/\delta_{\text{tot}}$, is in general a function of the scale, the properties of the galaxy in question, and the redshift. Numerical simulations complement the analytical results (as pioneered, e.g., by Kaiser (1984) and Bardeen (1986) [37]), and show that galaxies preferentially form in halos that are not too small (otherwise supernova feedback hampers the collapse) nor too large (otherwise gas does not cool efficiently enough).

Simulations of cold collision-less DM produce DM halos, which have in general an *ellipsoidal shape*, rather than being spherical. The non-sphericity is limited, however, and is found to be smaller in simulations that include baryons, as we will discuss in section 2.4.1. The standard assumption of sphericity, adopted in most phenomenological studies, is therefore quite reasonable.

The **density profiles** $\rho(r)$ of individual halos, computed from numerical data in spherical approximation, tend to follow a universal slope $s(r) \equiv d\ln\rho/d\ln r$ that smoothly changes from $\sim -1$ (in the inner region) to $\sim -3$ (in the outer region). For comparison, the isothermal sphere, suggested by the analytic considerations in eq. (2.8), has $s = -2$. The results of simulations can be fitted by functions such as the ones in table 2.1 dubbed 'NFW' (from Navarro, Frenk, White; with a slope that changes from $-3$ to $-1$ around the radius $r_s$) or 'Einasto' (with a slope that equals $-2$ at $r = r_s$ and continuously evolves). The numerical values of DM density profile fit functions for Milky Way will be discussed in section 2.2.2. The individual gravitationally bound objects can be simulated down to sizes about 3 orders of magnitude smaller than their virial radius $r_{\text{vir}}$ (defined such that the $\rho(r_{\text{vir}})$ is 200 times larger than the mean density, for reasons discussed in section 2.1.2). The simulations thus cannot resolve a possible core (where the density $\rho$ becomes constant), if it is too small. As a rule of thumb, for a galaxy like the Milky Way, the existence of a core with a size below $\approx 1$ kpc is compatible with the current simulations.

In addition to the primary halo structures, simulations reveal the presence of numerous smaller **sub-halos** (see fig. 2.2) orbiting within and around main halos. Their number distribution is $dn/d\mathcal{M} \gtrsim 1/\mathcal{M}^2$, so that $\mathcal{M}\, dn/d\ln\mathcal{M}$ is roughly $\mathcal{M}$-independent. That is, similar amounts of total DM mass are stored within objects for any given decade in mass scale $\mathcal{M}$, from the biggest formed structures down to the smallest resolved structures, of the order $\mathcal{M} \approx 10^5\ M_\odot$ [38]. Additional smaller halos, not yet resolved by the simulations, are expected to exist, as also anticipated by the analytic arguments to be discussed in section 2.2.5. Simulations show that these sub-halos possess higher inner DM densities and are, on average, more concentrated than stand-alone halos of the same mass. In section 6.8.1 we will discuss how the presence of sub-halos boosts the indirect signals of DM annihilations.

Other features of simulated halos seem less universal. The concentration parameter of any halo is defined as $c = r_{\text{vir}}/r_{-2}$ where $r_{-2}$ is the radius at which the slope reaches $s = -2$ ($r_{-2} = r_s$ in the Einasto parameterisation). The mass of the smallest halos, called **micro-halos**, will be discussed in section 2.2.5:

| DM halo | Functional form |
|---|---|
| NFW | $\rho_{\text{NFW}}(r) \;=\; \rho_s \dfrac{r_s}{r}\left(1+\dfrac{r}{r_s}\right)^{-2}$ |
| Generalized NFW | $\rho_{\text{gNFW}}(r) \;=\; \rho_s \left(\dfrac{r_s}{r}\right)^{\gamma}\left(1+\dfrac{r}{r_s}\right)^{\gamma-3}$ |
| Einasto | $\rho_{\text{Ein}}(r) \;=\; \rho_s \exp\left\{-\dfrac{2}{\alpha_{\text{Ein}}}\left[\left(\dfrac{r}{r_s}\right)^{\alpha_{\text{Ein}}}-1\right]\right\}$ |
| Cored Isothermal | $\rho_{\text{Iso}}(r) \;=\; \dfrac{\rho_s}{1+(r/r_s)^2}$ |
| Burkert | $\rho_{\text{Bur}}(r) \;=\; \dfrac{\rho_s}{(1+r/r_s)(1+(r/r_s)^2)}.$ |

Table 2.1: *Plausible spherical density profiles $\rho(r)$ for DM halos in galaxies.*

it depends on DM particle properties and could, for weak-scale DM, be comparable to Earth's mass, $10^{-6} M_\odot$. Micro-halos do not form from merging of smaller halos — dedicated simulations suggest that they tend to have $c \approx 20 - 30$ [39]. The concentration parameter $c$ gradually decreases down to $c \sim$ few for larger objects (galaxies and clusters), which formed earlier, when the universe was denser. Sub-halos inside galactic halos in particular have higher inner DM densities and are, on average, more concentrated than the free-standing halos of the same mass. Moreover, the sub-halos are more concentrated, the closer they are to the center of the host halo [38]. Finally, the ratio between the spherically averaged density and the average radial velocity seems to follow a power-law, $\rho/v_r^3 \propto 1/r^{1.875}$.

Simulations have intrinsic limitations: normal matter is only approximatively modeled, numerical issues remain, halos selected for more detailed computations are maybe not the typical ones, etc. Comparing simulations to observations suggest possible discrepancies, discussed in section 8.5: the cusp-core problem, the diversity problem, and the missing satellite problem.

The results of $N$-body simulations apply to the Milky Way, as long as this is a typical galaxy. Since we do not know observationally whether or not the Milky Way is surrounded by a halo with an atypically high asphericity, we will later assume that this is not the case, and use an average spherical DM halo to describe the Milky Way. This may well be an oversimplification: the measured Milky Way stellar velocities indicate that it formed via a relatively recent merging of two big progenitors [40, 41], possibly complicating the DM local distribution.

## 2.2 Dark Matter density distribution

### 2.2.1 Galactic DM density functions

We review now the galactic DM density distributions $\rho(r)$ often considered as the most plausible in the literature. All the profiles assume spherical symmetry, with $r$ the radial coordinate measured from the Galactic Center (GC). The functions are listed in table 2.1 and plotted in fig. 2.3 for plausible values of their free parameters $r_s$, $\rho_s$ and $\alpha_{\text{Ein}}$. Some of these functions can be conveniently recast in terms of a collective formula with three parameters $\alpha, \beta, \gamma$ [42], also referred to as the 'double power-law' formula

$$\text{Hernquist } \alpha\beta\gamma: \quad \rho_{\alpha\beta\gamma}(r) \;=\; \frac{\rho_s}{(r/r_s)^{\gamma}\left[1+(r/r_s)^{\alpha}\right]^{\frac{\beta-\gamma}{\alpha}}}, \qquad
\begin{array}{l|ccc}
 & \alpha & \beta & \gamma \\
\text{NFW} & 1 & 3 & 1 \\
\text{gNFW} & 1 & 3 & \gamma \\
\text{Cored Iso} & 2 & 2 & 0
\end{array} \quad . \tag{2.10}$$

The parameters $\alpha$, $\beta$, and $\gamma$ control the shape of the DM density profile at different radial distances from the galactic center. Specifically, $\alpha$ controls the sharpness of the transition between the inner and outer slopes, $\beta$ affects the outer slope, and $\gamma$ determines the inner slope of the profile. These functions are motivated by the following considerations:

    ▷ The Navarro, Frenk and White (**NFW**) [43] profile is the most common benchmark choice motivated by the $N$-body simulations. The density diverges as $r^{-1}$ close to the GC. The version with two parameters fixed, $\alpha = 1, \beta = 3$, and $\gamma$ taken as a free parameter (in the notation of eq. (2.10)), is sometimes called the '**generalized NFW (gNFW)**' profile. When $\gamma > 1$, the slope at the center is steeper than for the standard NFW: in this case the profile is referred to as the '**contracted NFW (cNFW)**'. For instance, this had been proposed by **Moore** and collaborators [44], who found $\gamma = 1.16$. Such contracted profiles emerged in some early numerical simulations which included baryons.

    ▷ The **Einasto** [45] profile emerged as a better fit to more recent numerical simulations. Loosely speaking, the Einasto profile density is more 'chubby' than NFW. More precisely, the Einasto power-law exponent continuously changes with $r$, as controlled by the shape parameter $\alpha_{\text{Ein}}$. The value $\alpha_{\text{Ein}} = 0.17$ represents a reasonable average over different values suggested by different $N$-body simulations.

    ▷ Cored profiles, such as the 'cored' ('truncated') **Isothermal** profile [46] or the **Burkert** profile [47], feature a constant central density. They have been motivated by some observations of galactic rotation curves that may be pointing toward the presence of a core, see, e.g., [48], and by numerical simulations finding that the effect of baryon feedback lowers the central density, see, e.g., [49].[6]

    ▷ Di Cintio et al. (2014) [50] suggested a profile, which has the double power-law form in eq. (2.10), but with parameters $\alpha, \beta, \gamma$ that are not the same across all galaxies, but rather depend on the stellar-to-halo mass ratio of each galaxy.[7] The DM profiles then effectively range from cusped to cored ones, adapting to the stellar and DM content of each galaxy. This is based on the results of simulations that include baryons, as discussed in section 8.5.1.

In some of the considered profiles $\rho(r)$ diverges as $r \to 0$, however, in all cases $r^2 \rho(r) \to 0$ (unless $\gamma$ is unrealistically large) such that the central region of the Galaxy contains just a small amount of DM.

While the various density profiles give similar results at distances larger than a few kpc from the GC, including around the location of the Earth, they do differ considerably — by orders of magnitudes — at smaller distances. Close to the GC there are no observational data on the DM profile. Moreover, the resolution of numerical simulations does not allow to go below, say, a fraction of a kpc. The behaviour of $\rho(r)$ is thus simply governed by the assumed asymptotic functional form as $r \to 0$. As a consequence, indirect DM signals from the inner Galaxy, such as gamma ray fluxes from regions a few degrees around the GC, depend strongly on the uncertain DM profile. This is in contrast to many other DM signals, which depend on DM profiles away from the GC: DM direct detection signals depend on the DM density at the position of the Earth; a number of DM signals probe the local Galactic neighbourhood (e.g., the fluxes of high energy positrons, produced at most a few kpc away from the Earth), as well as DM signals that probe regions distant from the GC (e.g., gamma rays from high latitudes).

## 2.2.2    Determination of the Milky Way parameters

Next, one has to determine the parameters $r_s$ (typical scale radius) and $\rho_s$ (typical scale density) that enter in the tentative DM distributions $\rho(r)$. This can be done in different ways, e.g., by extracting

---

    [6]The fact that, at face value, $N$-body numerical simulations differ from observations is known as the **cusp-core problem** and it is discussed more thoroughly in section 8.5.1.

    [7]See eq. (3) in Di Cintio et al. (2014) [50] for the explicit functional forms.

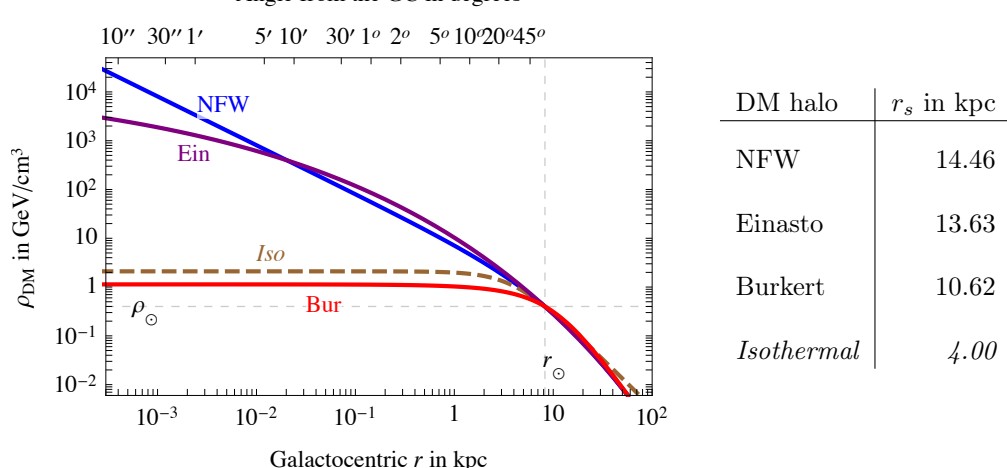

| DM halo | $r_s$ in kpc | $\rho_s$ in GeV/cm$^3$ |
|---|---|---|
| NFW | 14.46 | 0.566 |
| Einasto | 13.63 | 0.154 |
| Burkert | 10.62 | 1.143 |
| *Isothermal* | *4.00* | *2.112* |

Figure 2.3:  **DM profiles** *(figure left) and (table right) the corresponding parameters in the parametrizations of the profiles in Table 2.1. The procedure to determine the parameters of the Isothermal profile is different from the other ones, see the text for details. In the table we provide $r_s$ ($\rho_s$) to 2 (3) significant digits, a precision sufficient for most computations. Still more precise inputs are needed in specific cases, such as to precisely reproduce the J factors (discussed in section 6.2) for small angular regions around the Galactic Center.*

their values from numerical simulations of Milky Way-like halos, or determining them in some way from observations of the Milky Way or similar outer galaxies. One approach is to impose that the DM profiles for the Milky Way satisfy the following set of constraints:

A) The **density of Dark Matter at the location**[8] of the Sun $\rho_\odot$. This quantity was studied by many groups using a number of different techniques [51], notably the *global method* of fitting the entire rotation curve of the Galaxy, as discussed above, or the *local methods*, which rely on studying local stellar kinematics (especially the stellar motions in the vertical direction) to determine the local gravitational pull and therefore the local DM density. The global method can provide a precise determination of $\rho_\odot$, but is sensitive to the uncertain modeling of the baryonic components of the Galaxy. The local methods are less precise and suffer, on one hand, from the systematics due to peculiar local conditions (such as the asymmetries in the north and south galactic hemispheres) and, on the other hand, from the simplifications in the analysis (e.g., whether or not the so-called 'tilt term', which correlates radial and vertical stellar motions, is included).

The recent determinations of $\rho_\odot$ point towards

$$\rho_\odot = \rho(r_\odot) = 0.40 \text{ GeV/cm}^3 \approx 0.0106\,M_\odot/\text{pc}^3 \qquad (2.11)$$

The typical associated error is $\pm 0.10$ GeV/cm$^3$, but a possible spread up to $0.2 \leftrightarrows 0.8$ GeV/cm$^3$ (sometimes referred to as 'a factor of 2') is often adopted. As is evident from the selected results listed in table 2.2, different determinations of $\rho_\odot$ are still in poor agreement. Historically, the conventional value used in the literature since the 1990s used to be $\rho_\odot = 0.3$ GeV/cm$^3$. This value increased following several studies, performed mostly between 2009 and 2011, which found

---

[8]The distance of the Sun from the Galactic Center [52] is also somewhat uncertain. In recent years the central value has fluctuated around 8.3 kpc, with an error of about $\pm 0.3$ kpc. One of the most recent and most precise determinations, due to the GRAVITY collaboration [52], yields $r_\odot = 8.277 \pm 0.031$ kpc (statistical and systematic errors summed in quadrature).

| Authors | Date | $\rho_\odot$ in GeV/cm$^3$ | Notes |
|---|---|---|---|
| Turner | 1986 | 0.28 | 'uncertainty of about a factor of 2' |
| Flores | 1988 | $0.3 \leftrightarrow 0.43$ | |
| Kuijken & Gilmore | 1991 | $0.42\ (\pm 20\%)$ | |
| Widrow et al. | 2008 | $0.304 \pm 0.053$ | |
| Catena & Ullio | 2009 | $0.385 \pm 0.027$ | Einasto |
| | | $0.389 \pm 0.025$ | NFW |
| Weber & de Boer | 2009 | $0.2 \leftrightarrow 0.4$ | |
| Salucci et al. | 2010 | $0.43 \pm 0.11 \pm 0.10$ | |
| McMillan | 2011 | $0.40 \pm 0.04$ | |
| Garbari et al. | 2011 | $0.11^{+0.34}_{-0.27}$ | isothermal stellar tracers |
| | | $1.25^{+0.30}_{-0.34}$ | non-isothermal stellar tracers |
| Iocco, Pato & Bertone | 2011 | $0.2 \rightarrow 0.56$ | |
| Bovy & Tremaine | 2012 | $0.3 \pm 0.1$ | |
| Zhang et al. | 2012 | $0.28 \pm 0.08$ | |
| Piffl et al. | 2014 | $0.59\ (\pm 15\%)$ | |
| Pato, Iocco & Bertone | 2015 | $0.420 \pm 0.025$ | |
| McKee et al. | 2015 | $0.49 \pm 0.13$ | |
| McMillan | 2016 | $0.40 \pm 0.04$ | |
| Sivertsson et al. | 2017 | $0.46^{+0.07}_{-0.09}$ | |
| Buch et al. | 2018 | $0.608 \pm 0.380$ | result depends on chosen tracers |
| Eilers et al. | 2018 | $0.30 \pm 0.03$ | |
| Evans et al. | 2018 | $0.55 \pm 0.17$ | |
| Karukes et al. | 2019 | $0.43 \pm 0.02 \pm 0.01$ | stat and sys errors |
| Cautun et al. | 2020 | $0.33 \pm 0.02$ | |
| Sofue | 2020 | $0.359 \pm 0.017$ | own determination |
| | | $0.39 \pm 0.09$ | average of recent results |
| Salomon et al. | 2020 | $0.42 \leftrightarrow 0.53$ | |
| Hattori et al. | 2020 | $0.342 \pm 0.007$ | |
| Benito, Iocco & Cuoco | 2021 | $0.4 \leftrightarrow 0.7$ | inferred $2\sigma$ range |
| Lim, Putney, Buckley, Shih | 2023 | $0.446 \pm 0.054$ | |
| Staudt et al. | 2024 | $0.42 \pm 0.06$ | from simulations |

Table 2.2: ***DM density at the location of the Sun***. *The top set consists of 'historical' studies. The bottom set contains a selection of works published after the release of* GAIA *data (although not necessarily using them). The middle set lists selected results of the latest* $\sim 15$ *years. Note that the position of the Sun may not be exactly the same for all authors, that the methods are different, that the key assumptions might differ, and that relative renormalizations might be necessary. References are in [51].*

somewhat higher central values. Other studies, on the other hand, found smaller central values.[9] Additionally, the associated error remained a topic of intense debate. Data from the GAIA mission stimulated new activities aimed at determining $\rho_\odot$, but the situation is not yet settled, and a more precise determination of $\rho_\odot$ seems difficult to achieve.

For comparison, the total matter density (including normal matter) averaged over a neighborhood of a few hundred parsecs around the location of the Sun is estimated to be an order of magnitude larger than the DM density (2.11) [53],

$$4.5\,\text{GeV/cm}^3 \approx 0.120\,M_\odot/\text{pc}^3. \tag{2.12}$$

---

[9]In 2012 the result from Moni Bidin et al. [51] made the news, since they found $\rho_\odot = 0.000 \pm 0.004$ GeV/cm$^3$, later ascribed to poor modeling.

Eq. (2.11) means that the mass of the Dark Matter included within the orbit of Neptune ($\sim 30$ AU) is of the order of $10^{-13} M_\odot$ or, equivalently, that the mass of the Sun corresponds to the mass of all the Dark Matter contained in a sphere of radius $2.8\,\mathrm{pc} \approx 9.2\,\mathrm{ly}$, comparable to the distance to the nearest star, about 4 ly. This illustrates how DM is largely subdominant locally. One may wonder whether its presence in the solar system could nevertheless be detected via its influence on the orbits of planets and the closely monitored asteroids (for one, the accumulated DM would slightly increase the effective mass of the Sun as seen by the outer bodies). However, direct astronomical observations, while accurate, are still not particularly constraining. The resulting bound on DM density in the solar system is $\rho \lesssim 5000\,\mathrm{GeV/cm^3}$, which is about 15000 times larger than the expected $\rho_\odot$ [54]. In other words, DM has no significant effect on the orbits of the planets in the solar system.

A related issue is whether the average value for $\rho_\odot$ in eq. (2.11) can be modified significantly by the presence of astrophysical bodies in the solar system, e.g., whether it could get increased by the gravitational attraction of the Sun or decreased by slingshot effects acting on DM particles passing the planets (notably Jupiter, but also Venus and the Earth itself, in the same way as what happens for conventional asteroids). Any appreciable modification of local DM density and speed distributions would be particularly important for the interpretations of DM direct detection results (see chapter 5), and for predicting the neutrino flux from DM particles captured in the Sun or the Earth (see section 6.9). A series of detailed studies identified a number of subtle effects that, for the most part, cancel each other out, leading to a practically negligible net effect, at least for DM particles that interact only weakly with the astrophysical bodies [55].

Note also that the energy density in DM is $\rho_\odot \sim 20\,\mathrm{kW/m^2}$,[10] about 14 times higher than the power density of the Sun's radiation on the surface of the Earth (good to keep in mind, just in case this energy someday turns out to be in a usable form, converting our universe into one giant battery pack).

B) The **total DM mass contained in the Milky Way**. This quantity was also studied by many groups, using different techniques [56]. The methods include the study of the kinematics of bright satellites (such as the dwarf galaxies or the Magellanic clouds), the kinematics of stellar streams, the escape velocity of nearby rogue stars and the use of various tracers. For $M_{200}^{\mathrm{DM}}$, the mass of Dark Matter contained inside a virial radius $r_{200}$, defined in section 2.1.2 (below eq. (2.3)), we will use the 'standard' value

$$\boxed{M_{200}^{\mathrm{DM}} \equiv (1.0 \pm 0.25) \times 10^{12}\, M_\odot}. \tag{2.13}$$

However, it is worth noting that the advent of the GAIA mission and its precise astrometric data resulted in a lot of activity recently, with a number of studies in 2023 arriving at a factor of a few lower value for $M_{200}^{\mathrm{DM}}$ [7]. For the Milky Way, $r_{200} \sim 200$ to $300$ kpc, hence one can think of $M_{200}^{\mathrm{DM}}$ as the total DM mass contained in a large sphere of radius 200 kpc centered around the center of the Galaxy.

For comparison, the stellar mass inferred with similar techniques is $\approx 8 \times 10^{10}\, M_\odot$. The distance from the Earth to the Large Magellanic Cloud, is about 48.5 kpc.

Imposing the constraints (2.11) and (2.13) fixes the $r_s$ and $\rho_s$ parameters[11] for the NFW, Einasto and Burkert profiles in table 2.1. For the Isothermal profile, however, no solution can be found that would

---

[10]Using $c = 1$ units. One can think of it as the energy flux seen by an observer moving at the speed of light, if all DM rest mass were to be absorbed. Since DM interacts only feebly with matter (if at all), the probability of DM actually being absorbed is miniscule, making it exceedingly hard to tap into this source of energy.

[11]We should stress that the different parameters discussed in these sections ($\rho_\odot, r_\odot, M_{200}^{\mathrm{DM}}$ and, later on, the rotational velocity of the Sun) are interconnected and correlated. As discussed above, they are typically determined within a global galactic model, which consists of the baryonic components but often also requires to specify a DM density distribution and a DM velocity distribution. Hence, strictly speaking, using the values $\rho_\odot, r_\odot$ and $M_{200}^{\mathrm{DM}}$ to constrain the DM distributions is inconsistent. However, the impact of the choice of the DM distributions on the global model is limited, thus the procedure can be considered as valid for all current practical purposes.

simultaneously satisfy both constraints: the $r^{-2}$ behavior of DM density at large $r$ is too slow and always results in $M_{200}^{\mathrm{DM}}$ larger than the value in eq. (2.13). For this profile we therefore disregard the constraint in eq. (2.13); instead we set $r_s = 4$ kpc, as was done in the original literature, and then determine the value of $\rho_s$ for which the constraint on the local density in eq. (2.11) is satisfied. The results of this procedure are presented in fig. 2.3, where we also plot the resulting profiles. The values of the parameters determined with this procedure, and presented here, in general do not differ much, at most by 20%, from the parameter values often used in the literature.

### 2.2.3   Dwarf galaxies

The Milky Way is surrounded by a number of *satellite galaxies* [57], i.e., astrophysical systems that are tied to the main MW halo by gravitational attraction. Apart from the Large and Small Magellanic Clouds, which are big enough to be seen with a naked eye, the large majority of the satellites consists of *dwarf galaxies*. Most of these are of the *dwarf spheroidal* (dSph) type and only a few are dwarf irregulars (dIrr). Historically, Sculptor was the first one to be discovered (in 1938) and it is often considered as the prototypical dSph. Seven more dwarf galaxies (Fornax, Leo I, Leo II, Ursa Minor, Draco, Carina and Sextans) were discovered up to the 1990s. Collectively, these 8 dwarf galaxies are referred to as *classical*. With the advent of the digital surveys such as Sdss, the Subaru Hyper-Supreme Camera (Hsc) (which covered the northern sky), Pan-Starrs, Des (which covered the southern sky) and Gaia (operating from space), many more dwarfs have been discovered: these are referred to as *ultrafaint*. The current total stands just short of 50, including several whose nature and properties are not yet confirmed (see table 2.3).[12] The list is probably going to increase, as the surveys become more sensitive and cover larger portions of the sky, e.g., with the upcoming Rubin Observatory (formerly known as Lsst). There is also the Sagittarius dSph, discovered in 1994, which is nearby and large, but stands apart, since it is believed to be tidally disrupted by the MW halo.

The definition of the 'dwarf' category is not clear cut, i.e., there are no universally accepted ranges of values for the mass, the size, and the luminosity of a small galaxy that would define it as being of the dwarf type. In general, one can think of a dwarf galaxy as having a DM-dominated mass of around $10^6 \, M_\odot$ (see fig. 3.1), where the actual dwarf galaxies can differ by a couple of orders of magnitude from this value: for instance, Fornax is estimated at $56 \cdot 10^6 \, M_\odot$ while Segue 2 at $0.23 \cdot 10^6 \, M_\odot$ [57]. The half-light radius $r_{\mathrm{HL}}$, defined as the radius of a circle[13] that contains half of the light emission from the galaxy, can be from a few tens of parsecs to several hundred parsecs. According to simulations, the virial radius $r_{200}$ of the DM halo of the dwarf galaxy tends to be one order of magnitude bigger than $r_{\mathrm{HL}}$ [60]. The number of stars identified as belonging to the dwarf galaxy can be as low as a dozen or as many as several thousand.

Dwarf galaxies are promising targets for DM searches, because they contain only a few stars and are believed to be dominated by Dark Matter. The likely reason is that dwarf galaxies constitute relatively shallow potential wells, compared to larger galaxies, facilitating the expulsion of baryonic gas by early supernovæ, while the subsequent star formation rate is low. The estimated mass-to-light ratios $\mathcal{M}_{\mathrm{tot}}/L$ can be of the order of several tens $M_\odot/L_\odot$ (e.g., Sculptor) to hundreds $M_\odot/L_\odot$ (e.g., Draco), and can even reach a few thousand $M_\odot/L_\odot$ (e.g., Segue 1).[14] The large DM content of dwarfs is of course good news for the expected intensity of the DM signal. Moreover, the scarcity of stars and other active astrophysical

---

[12] Xu et al. (2023) [59] claim a discovery of the first DM-dominated dwarf galaxy external to the MW system. Fast J0139+4328 is a (very large) dwarf galaxy located at about 29 Mpc which is found to have a DM-dominated ratio of total mass vs baryonic mass, $47 \pm 27$.

[13]For objects with a very elliptical shape, a 'half-light ellipse' with different major and minor axes is sometimes defined. Spherical symmetry is however usually assumed for simplicity.

[14]Here, $L_\odot = 3.828 \cdot 10^{26}$ W is the nominal luminosity of the Sun, while $M_\odot = 1.9984 \cdot 10^{30}$ kg is the already encountered solar mass.

| Name | Discovery | Position $(b, \ell)$ | Distance in kpc | $r_{\rm HL}$ in pc | $\log_{10}\mathcal{J}_{\rm dSph}({\rm disk}_{0.5°})$ in $\mathrm{GeV^2\,cm^{-5}}$ | $\log_{10}\mathcal{D}_{\rm dSph}({\rm disk}_{0.5°})$ in $\mathrm{GeV\,cm^{-2}}$ |
|---|---|---|---|---|---|---|
| Sculptor | 1938 | $(-83.2°, 287.5°)$ | $86 \pm 6$ | $283 \pm 45$ | $18.58^{+0.05}_{-0.05}$ | $18.20^{+0.09}_{-0.08}$ |
| Fornax | 1938 | $(-65.7°, 237.1°)$ | $147 \pm 12$ | $710 \pm 77$ | $18.09^{+0.10}_{-0.10}$ | $17.97^{+0.05}_{-0.05}$ |
| Leo I | 1950 | $(+49.1°, 226.0°)$ | $254 \pm 15$ | $251 \pm 27$ | $17.61^{+0.13}_{-0.11}$ | $18.01^{+0.20}_{-0.28}$ |
| Leo II | 1950 | $(+67.2°, 220.2°)$ | $233 \pm 14$ | $176 \pm 42$ | $17.66^{+0.16}_{-0.15}$ | $17.64^{+0.50}_{-0.33}$ |
| Ursa Minor | 1955 | $(+44.8°, 105.0°)$ | $76 \pm 3$ | $181 \pm 27$ | $18.76^{+0.12}_{-0.11}$ | $18.20^{+0.14}_{-0.08}$ |
| Draco | 1955 | $(+34.7°, 86.4°)$ | $76 \pm 6$ | $221 \pm 19$ | $18.82^{+0.12}_{-0.12}$ | $18.54^{+0.11}_{-0.14}$ |
| Carina | 1977 | $(-22.2°, 260.1°)$ | $105 \pm 6$ | $250 \pm 39$ | $17.83^{+0.09}_{-0.09}$ | $17.88^{+0.18}_{-0.15}$ |
| Sextans | 1990 | $(+42.3°, 243.5°)$ | $86 \pm 4$ | $695 \pm 44$ | $17.75^{+0.12}_{-0.11}$ | $17.90^{+0.11}_{-0.09}$ |
| Sagittarius | 1994 | $(-14.2°, 5.6°)$ | $26 \pm 2$ | $2587 \pm 219$ | $19.77^{+0.08}_{-0.07}$ * | $-$ |
| Willman 1 | 2005 (SDSS) | $(+56.8°, 158.6°)$ | $38 \pm 7$ | $25 \pm 6$ | $19.36^{+0.52}_{-0.46}$ | $18.52^{+0.47}_{-0.68}$ |
| Ursa Major I | 2005 (SDSS) | $(+54.4°, 159.4°)$ | $97 \pm 4$ | $319 \pm 50$ | $18.33^{+0.28}_{-0.28}$ * | $18.10^{+0.28}_{-0.29}$ * |
| Ursa Major II | 2006 (SDSS) | $(-37.4°, 152.5°)$ | $32 \pm 4$ | $149 \pm 21$ | $19.44^{+0.41}_{-0.39}$ | $18.64^{+0.28}_{-0.31}$ |
| Boötes I | 2006 (SDSS) | $(+69.6°, 358.1°)$ | $66 \pm 2$ | $242 \pm 21$ | $18.19^{+0.30}_{-0.28}$ | $18.11^{+0.25}_{-0.30}$ |
| Canes Venatici I | 2006 (SDSS) | $(+79.8°, 74.3°)$ | $218 \pm 10$ | $564 \pm 36$ | $17.43^{+0.16}_{-0.15}$ | $17.79^{+0.26}_{-0.27}$ |
| Canes Venatici II | 2006 (SDSS) | $(+82.7°, 113.6°)$ | $160 \pm 4$ | $74 \pm 14$ | $17.82^{+0.48}_{-0.47}$ | $18.01^{+0.36}_{-0.51}$ * |
| Coma Berenices | 2007 (SDSS) | $(+83.6°, 241.9°)$ | $44 \pm 4$ | $77 \pm 10$ | $19.01^{+0.36}_{-0.36}$ | $18.45^{+0.29}_{-0.44}$ |
| Hercules | 2007 (SDSS) | $(+36.9°, 28.7°)$ | $132 \pm 12$ | $330^{+75}_{-52}$ | $17.29^{+0.51}_{-0.52}$ * | $17.52^{+0.50}_{-0.51}$ * |
| Leo IV | 2007 (SDSS) | $(+57.4°, 264.4°)$ | $154 \pm 6$ | $206 \pm 37$ | $16.40^{+1.01}_{-1.15}$ * | $16.74^{+0.55}_{-0.56}$ * |
| Segue 1 | 2007 (SDSS) | $(+50.4°, 220.5°)$ | $23 \pm 2$ | $29^{+8}_{-5}$ | $19.00^{+0.48}_{-0.68}$ * | $18.08^{+0.53}_{-0.49}$ * |
| Leo T | 2007 (SDSS) | $(+43.7°, 214.9°)$ | $417 \pm 19$ | $120 \pm 9$ | $17.49^{+0.49}_{-0.45}$ | $17.67^{+0.53}_{-0.60}$ |
| Leo V | 2008 (SDSS) | $(+58.5°, 261.9°)$ | $178 \pm 10$ | $135 \pm 32$ | $17.65^{+0.91}_{-1.03}$ * | $17.13^{+0.65}_{-0.72}$ |
| Segue 2 | 2009 (SDSS) | $(-38.1°, 149.4°)$ | $35 \pm 2$ | $35 \pm 3$ | $16.21^{+1.08}_{-0.98}$ * | $15.89^{+0.56}_{-0.37}$ * |
| Sagittarius II | 2015 (Pan-STARRS) | $(-22.9°, 18.9°)$ | $67 \pm 5$ | $38^{+8}_{-7}$ | $17.35^{+1.36}_{-0.91}$ * | $-$ |
| Reticulum II | 2015 (DES) | $(-49.7°, 266.3°)$ | $31.4 \pm 1.4$ | $58 \pm 4$ | $18.94^{+0.38}_{-0.38}$ * | $18.30^{+0.33}_{-0.50}$ * |
| Eridanus II | 2015 (DES) | $(-51.6°, 249.8°)$ | $380$ | $158$ | $17.28^{+0.34}_{-0.31}$ | $17.60^{+0.42}_{-0.54}$ |
| Tucana II | 2015 (DES) | $(-52.3°, 328.1°)$ | $57 \pm 5$ | $165^{+28}_{-19}$ | $18.93^{+0.56}_{-0.50}$ | $18.46^{+0.35}_{-0.36}$ |
| Horologium I | 2015 (DES) | $(-54.7°, 271.4°)$ | $79$ | $31$ | $19.17^{+0.80}_{-0.70}$ | $18.14^{+0.65}_{-0.63}$ |
| Phoenix II | 2015 (DES) | $(-59.7°, 323.7°)$ | $84.1 \pm 8$ | $37 \pm 8$ | $18.20$ | $-$ |
| Tucana IV | 2015 (DES) | $(-55.3°, 313.3°)$ | $48 \pm 4$ | $127 \pm 24$ | $(18.7)$ | $-$ |
| Aquarius II | 2016 (SDSS) | $(-53.0°, 55.1°)$ | $108$ | $125$ | $18.39^{+0.65}_{-0.58}$ | $18.07^{+0.47}_{-0.50}$ |
| Crater II | 2016 (ATLAS) | $(+42.2°, 282.9°)$ | $117 \pm 1$ | $1066 \pm 84$ | $-$ | $-$ |
| Carina II | 2018 (MagLiteS) | $(-17.1°, 269.9°)$ | $36$ | $77$ | $18.37^{+0.54}_{-0.52}$ | $17.95^{+0.38}_{-0.40}$ |
| Carina III | 2018 (MagLiteS) | $(-16.8°, 270.0°)$ | $28$ | $20$ | $20.2^{+1.0}_{-0.9}$ | $18.8^{+0.6}_{-0.7}$ |
| Antlia II | 2019 (Gaia) | $(+11.2°, 264.9°)$ | $132$ | $2301$ | $-$ | $-$ |
| Pegasus IV | 2022 (DELVE) | $(-21.4°, 80.8°)$ | $90^{+4}_{-6}$ | $41^{+8}_{-8}$ | $-$ | $-$ |
| Virgo II | 2022 (DELVE) | $(+52.8°, 4.1°)$ | $72^{+8}_{-7}$ | $16 \pm 3$ | $-$ | $-$ |
| Boötes V | 2022 (DELVE/UNIONS) | $(+70.9°, 55.7°)$ | $102 \pm 21$ | $20 \pm 7$ | $-$ | $-$ |
| Leo Minor I | 2022 (DELVE) | $(+64.7°, 202.2°)$ | $82^{+4}_{-7}$ | $26 \pm 9$ | $-$ | $-$ |
| Triangulum II | 2015 (Pan-STARRS) | $(-23.4°, 141.4°)$ | $30 \pm 2$ | $28 \pm 8$ | $19.35^{+0.37}_{-0.37}$ * | $18.42^{+0.86}_{-0.79}$ * |
| Draco II | 2015 (Pan-STARRS) | $(+42.9°, 98.3°)$ | $20 \pm 3$ | $19^{+8}_{-6}$ | $18.93^{+1.39}_{-1.70}$ * | $18.02^{+0.84}_{-0.87}$ * |
| Pictor I | 2015 (DES) | $(-40.6°, 257.3°)$ | $114$ | $18$ | $-$ | $-$ |
| Grus I | 2015 (DES) | $(-58.2°, 338.7°)$ | $120$ | $21$ | $16.88^{+1.51}_{-1.66}$ | $17.00^{+0.87}_{-0.86}$ |
| Grus II | 2015 (DES) | $(-51.9°, 351.1°)$ | $53 \pm 5$ | $93 \pm 14$ | $(18.7)$ | $-$ |
| Tucana III | 2015 (DES) | $(-56.2°, 315.4°)$ | $25 \pm 2$ | $44 \pm 6$ | $(19.4)$ | $-$ |
| Columba I | 2015 (DES) | $(-28.9°, 231.6°)$ | $183$ | $98$ | $-$ | $-$ |
| Reticulum III | 2015 (DES) | $(-45.6°, 273.9°)$ | $92$ | $64$ | $-$ | $-$ |
| Tucana V | 2015 (DES) | $(-51.9°, 316.3°)$ | $55$ | $16$ | $-$ | $-$ |
| Indus II | 2015 (DES) | $(-37.4°, 354.0°)$ | $214$ | $180$ | $-$ | $-$ |
| Cetus II | 2015 (DES) | $(-78.5°, 156.5°)$ | $30$ | $17$ | $-$ | $-$ |
| Virgo I | 2016 (HSC) | $(+59.6°, 276.9°)$ | $91^{+9}_{-4}$ | $47^{+19}_{-13}$ | $-$ | $-$ |
| Cetus III | 2018 (HSC) | $(-61.1°, 163.8°)$ | $251^{+24}_{-11}$ | $90^{+42}_{-17}$ | $-$ | $-$ |
| Boötes IV | 2018 (HSC) | $(+53.3°, 70.7°)$ | $209^{+20}_{-18}$ | $462^{+98}_{-84}$ | $-$ | $-$ |
| Centaurus I | 2019 (DELVE) | $(+21.9°, 300.3°)$ | $116.3^{+1.6}_{-0.6}$ | $79^{+14}_{-10}$ | $-$ | $-$ |
| Eridanus IV | 2021 (DELVE) | $(-27.8°, 209.5°)$ | $76.6^{+4.0}_{-6.1}$ | $75^{+16}_{-13}$ | $-$ | $-$ |
| Ursa Major III | 2023 (UNIONS) | $(+73.7°, 194.6°)$ | $10 \pm 1$ | $3 \pm 1$ | $20.87^{+0.60}_{-0.58}$ * | $-$ |

Left-margin group labels: *classical* (Sculptor–Sextans), *confirmed ultrafaint* (Willman 1–Leo Minor I), *candidates* (Triangulum II–Ursa Major III).

Table 2.3: *Most of the discovered* **dwarf spheroidal satellite galaxies** *of the Milky Way. Data are from many different sources* [57, 58], *and not necessarily with homogenous methods. The* $\mathcal{J}_{\rm dSph}$ *and* $\mathcal{D}_{\rm dSph}$ *factors are defined in eq.* (6.8) *(*$\mathcal{D}$ *is proportional to the dwarf's mass due to DM) and are computed for a* $0.5°$ *disk. The cases with particularly large variations in the literature (*$\mathcal{O}(10)$ *in the central value) are labeled with* *. *The values in parenthesis are estimates.*

objects implies that the signal is less contaminated by foreground emissions compared to other systems. However, this comes at the price of having only a small number of tracers with which one can reconstruct the DM content and the DM density profile. Also, these tracers are typically only stars: no hydrogen clouds, such as those that crucially helped in tracing the rotation curve of spiral galaxies (see section 1.1), are found in these systems.

Because of stellar faintness and complex dynamics, the determinations of DM **density functions** in dSphs are challenging and subject to intense debate. For example, observationally there appears to be no consensus in the literature on whether the stellar kinematical data point to a cuspy or to a cored DM distribution. A number of measurements, especially in the early 2000's, had suggested the preference for a core in the centers of dSphs. Subsequent observations showed, at least in the specific cases such as Sculptor and a few others, that a cusped NFW profile also fits the data well. The results of numerical simulations, on the other hand, seem to support the formation of cores for middle-sized dwarfs, while low mass dwarfs appear to develop cusps (consistent with the trend discussed in section 1.3.2). The core-vs-cusp nature of dwarf galaxies might also depend on the history of star formation: dwarfs that stopped forming stars more than 6 Gyrs ago developed cusps by unimpeded contraction, while those that kept forming stars have had their DM content 'heated' by the stellar bursts and have therefore developed cores.

A common practical approach in the literature involves using profiles like those discussed in section 2.2, with parameters determined in a number of different ways (see, e.g., Geringer-Sameth (2015) in [58]). From these one then predicts the corresponding DM signals (see section 6.2.2).

## 2.2.4 Galaxy clusters

Galaxy clusters are the largest gravitationally bound systems in the Universe. They contain hundreds to thousands of galaxies, extend to several Mpc in size and have typical total masses up to $10^{14} - \text{few } 10^{15} \, M_\odot$. Galaxy clusters can be considered, in some respect, as a scaled-up version of galaxies, which provides a basic framework for understanding their principal characteristics (for detailed reviews see [61]). However, significant differences also exist between galaxy clusters and galaxies.

Clusters contain baryonic and dark matter in proportions that are typically 1 to 10 (see section 1.2). The formation of the DM halo of a cluster can be described, in first approximation, as the non-linear spherical collapse of an initial over-density, in a way which is entirely analogous to that described in section 2.1.2 at the galactic scale. For more precise modeling, numerical simulations, which may include only dark matter or both dark matter and baryons, are employed. These simulations show that the NFW and Einasto profiles introduced in section 2.2 are good fits for the cluster DM distribution. This stresses that such profiles are roughly *universal* functions, i.e. they apply to collapsed systems independently of their mass. Like for galaxies, lensing observations seem however to show that the slope of the density profile in central regions of some clusters is shallower than predicted.

Perhaps the most important qualitative difference between galaxies and galaxy clusters is that most baryons in clusters (as in the Universe) are in the form of diffuse gas, not stars. This implies a limited impact of baryonic physics in shaping the DM distribution: gas essentially only cools, and cooling does not have a major effect on the profiles (except very close to the inner regions of the cluster).

## 2.2.5 The halo mass function

In this section we compute the *halo mass function $dn(t)/d\mathcal{M}$*: the average cosmological number density $n$ of gravitationally bound structures (galaxies, clusters of galaxies, dark matter halos, etc) with mass $\mathcal{M}$, at a given time $t$. Determining this quantity is interesting *per se*, since it is a fundamental prediction of the theory of structure formation. At the same time, the knowledge of $dn(t)/d\mathcal{M}$ is also necessary for predicting the cosmological signals of DM annihilations.

The halo mass function $dn(t)/d\mathcal{M}$ can be most accurately extracted from the numerical $N$-body simulations (sections 1.3.2, 2.1.3). Historically, however, it has been determined using approximate analytic treatments [62], which, it turns out, reproduce surprisingly well the numerical results, despite some questionable assumptions and drastic simplifications. In the following, we review the analytical derivations, and mention how they have been improved in order to better agree with the simulations.

In section 1.3.1 we reviewed the analytical theory of structure formation in the limit $\delta(\boldsymbol{x}, t) \ll 1$, where $\delta$ is the density contrast. In this regime the Universe is still nearly homogeneous. Focusing on the period after matter/radiation equality (which is the one relevant for structure formation), all the modes of density perturbations that are well inside the horizon evolve in the same way, so that $\delta(\boldsymbol{x}, t) = D(t/t')\delta(\boldsymbol{x}, t')$, where $\boldsymbol{x}$ is the comoving coordinate and $D$ is the so called *growth factor*. Below, we extrapolate these results to later times, when structures start to form. This may be questionable, since the gravitationally bound objects start to form when $\delta \sim 1$, i.e., in the mildly non-linear regime. More precisely, both the analytical computations and the numerical simulations show that the structures form when $\delta > \delta_c$, where $\delta_c$ is the critical value of density contrast, equal to $\delta_c \approx 1.686$ for a spherically symmetric case. Extrapolating the small-$\delta$ regime up to $\delta \sim \delta_c$ is good enough for our purposes, since $\delta_c$ is around unity.

We will assume that the density contrast $\delta$ follows a Gaussian distribution, see the discussion in the introduction to section 1. (In reality, $\delta \geq -1$, because density is positive, and thus the distribution needs to be truncated.) The variance of the density contrast distribution can be expressed as

$$\sigma^2 = \frac{\int d^3x\, \delta^2(\boldsymbol{x})}{\int d^3x} = \int \frac{d^3k}{(2\pi)^3} P(k) = \int \frac{dk}{k} \Delta^2(k), \tag{2.14}$$

where $\Delta^2(k) = k^3 P(k)/2\pi^2$ is a dimension-less variance, while the power spectrum $P(k)$ was introduced in section 1.3.1. The variance $\sigma^2$ diverges, because the fluctuations are dominated by small scales, i.e., large $k$. This is cured by replacing $\delta(\boldsymbol{x})$ with $\delta_R(\boldsymbol{x})$, the density contrast smoothed over scales of order $R$. To perform the smoothing we introduce a 'window function' $W_R(\boldsymbol{x})$, which is sizable only within roughly a distance $R$ from the origin. This is then convoluted with $\delta$ to obtain:

$$\delta_R(\boldsymbol{x}) = \int d^3x'\, \delta(\boldsymbol{x}')W_R(\boldsymbol{x} - \boldsymbol{x}'), \qquad \text{i.e.,} \qquad \delta_{Rk} = W(kR)\delta_k. \tag{2.15}$$

Here, $W(kR) = \int d^3x\, W_R(\boldsymbol{x}) \exp(-i\boldsymbol{k} \cdot \boldsymbol{x})$ is the Fourier transform of $W_R(\boldsymbol{x})$. The window function is normalised as $\int d^3x\, W_R(\boldsymbol{x}) = 1$. Common choices are a sharp cut in $\boldsymbol{k}$ space ($W(kr) = 0$ outside a sphere $k < 1/R$ and constant inside), or a sharp cut in $\boldsymbol{x}$ space ($W_R(\boldsymbol{x}) = 0$ outside $x > R$ and constant inside); or a Gaussian function in both spaces ($W_R \propto e^{-r^2/2R^2}$, $W = e^{-k^2R^2/2}$).

The variance of the smoothed inhomogeneity, computed analogously to eq. (2.14),

$$\sigma_R^2(t) = \int \frac{dk}{k} \Delta^2(k, t)W^2(kR), \tag{2.16}$$

is finite and increases at small $R$. This smoothing has the side effect of counting as collapsed any under-dense regions that are located inside collapsed over-densities.

Next, following Press and Schechter (PS) [62], we somewhat questionably adopt the following ansatz: the fraction $\mathcal{F}$ of the volume of the Universe that, at time $t$, has collapsed and formed structures with radius $R$, is *twice* the volume in which $\delta > \delta_c$. The latter is equal to the probability that $\delta > \delta_c$. Since $\delta$ has been taken as Gaussian-distributed with variance $\sigma_R^2$, we therefore have

$$\mathcal{F}(R) = p(\delta > \delta_c|R) = \frac{2}{\sqrt{2\pi}\sigma_R} \int_{\delta_c}^{\infty} d\delta\, e^{-\delta^2/2\sigma_R^2} = \text{erfc}\left[\frac{\delta_c}{\sqrt{2}\sigma_R}\right]. \tag{2.17}$$

The factor of 2 was added ad hoc, such that all the volume (as opposite to half the volume) is collapsed in the limit of large inhomogeneities, see Bond et al. (1991) in [62] for a more detailed explanation of this factor.

Finally, we obtain the halo mass function. We convert from volume to mass by estimating that a radius $R$ corresponds roughly to an enclosed mass $\mathcal{M} = 4\pi R^3 \bar{\rho}/3$, where $\bar{\rho}(t)$ is the average DM density, and by identifying the variance in mass with the variance in radius at the corresponding mass: $\sigma_{\mathcal{M}} = \sigma_{R(\mathcal{M})}$. The fraction of total mass that is gravitationally bound in structures with mass between $\mathcal{M}$ and $\mathcal{M} + d\mathcal{M}$ is given by $d\mathcal{F} = (d\mathcal{F}/d\mathcal{M})d\mathcal{M}$. The number of objects with mass between $\mathcal{M}$ and $\mathcal{M} + d\mathcal{M}$ is then obtained by multiplying the above result with the average number density $\bar{\rho}/\mathcal{M}$ of such objects: $dn/d\mathcal{M} = (\bar{\rho}/\mathcal{M})\, d\mathcal{F}/d\mathcal{M}$. Thereby the halo mass function is

$$\frac{dn}{d\mathcal{M}} = \frac{\bar{\rho}}{\mathcal{M}^2} f_{\mathrm{PS}}(\nu) \left| \frac{d\ln \sigma_{\mathcal{M}}}{d\ln \mathcal{M}} \right|, \qquad \text{where} \qquad f_{\mathrm{PS}}(\nu) = \sqrt{\frac{2}{\pi}} \nu e^{-\nu^2/2} \qquad \text{and} \qquad \nu \equiv \frac{\delta_c}{\sigma_{\mathcal{M}}(t)}. \qquad (2.18)$$

The number density $dn/d\mathcal{M}$ scales for small masses $\mathcal{M}$ roughly as $1/\mathcal{M}^2$, in agreement with the $N$-body simulations. At masses large enough such that $\sigma_{\mathcal{M}} \gtrsim \delta_c$, $dn/d\mathcal{M}$ gets roughly exponentially suppressed as $e^{-\nu^2/2}$.

Numerical simulations give results that mildly deviate from the Press-Schechter approximation: they predict more heavy halos, and fewer small halos. This is understood to be due to the fact that the simulated collapses tend to be ellipsoidal rather than spherical. When the spherical approximation is relaxed, the constant value for the typical critical over-density gets replaced by a more complicated $\mathcal{M}$ dependent function: $\delta_{\mathrm{ec}} \approx \delta_c \left[1 + 0.47(\sigma_{\mathcal{M}}^2/\delta_c^2)^{0.615}\right]$, see Sheth, Mo and Tormen (2001) in [62]. Here, the subscript $_{\mathrm{ec}}$ stands for *ellipsoidal collapse*. Taking into account the $\mathcal{M}$ dependence of $\delta_{\mathrm{ec}}$ provides an improved halo mass function, with $f_{\mathrm{PS}}(\nu)$ replaced by

$$f_{\mathrm{ec}}(\nu) \approx 0.3222 \, f_{\mathrm{PS}}(0.84\,\nu)[1 + (0.84\,\nu)^{-0.6}]. \qquad (2.19)$$

The main message of the halo mass function, either the one in (2.18) or in (2.19), is that halos exist at all scales. This implies, in particular, the existence of sub-halos within bigger galactic halos. One important consequence is the enhancement of DM annihilations within such sub-halos, which will be discussed in section 6.8.1.

Eq. (2.18) emphasises that the Press-Schechter formalism holds at any cosmological time $t$, providing an analytic quantitative understanding of the physics already described in section 1.3.2. Smaller $1/k$ scales re-entered the cosmological horizon earlier, so that $\delta_k(t)$ is larger for larger $k$. This means that DM started forming many small halos with density $\rho_{\mathrm{vir}} \sim \bar{\rho}$, i.e., comparable to the cosmological density $\bar{\rho}(t)$ at the formation time, as discussed around eq. (2.3). That discussion also showed that $\mathcal{M} \sim \rho_{\mathrm{vir}} r_{\mathrm{vir}}^3$, where $r_{\mathrm{vir}}$ is the radius of such halos, and that the virial velocity of DM is given by

$$v_{\mathrm{vir}} = \sqrt{\langle v^2 \rangle} \approx \sqrt{G\mathcal{M}/r_{\mathrm{vir}}} \sim r_{\mathrm{vir}}/r_{\mathrm{hor}} \sim \delta^{1/2}, \qquad (2.20)$$

where $r_{\mathrm{hor}} \sim 1/H \sim 1/\sqrt{G\bar{\rho}}$ is the horizon radius at the formation time. This shows that $v_{\mathrm{vir}}$ is non-relativistic (avoiding BH formation), because the inhomogeneities grew as in eq. (1.18), starting from the primordial value $\delta \approx 2\,10^{-5}$.

Ignoring interactions of normal matter, the Press-Schechter formalism accounts for the process of initially small halos merging into bigger ones, until the vacuum energy density started to dominate the cosmology and stopped structure formation. Since our focus is the DM, we will only briefly outline how normal matter ultimately leads to the visible structures we observe today. When normal matter falls into the DM gravitational potential well, it acquires temperature $T \sim m_p v_{\mathrm{vir}}^2$. The temperature $T$ controls, along with the density of normal matter, the rate at which various atomic processes dissipate energy. The denser halos, in which the rate of energy dissipation is faster than the (inverse of the) cosmological

Hubble time, form galactic disks and the first-generation stars. The halos in which the escape velocity is $v_{\text{vir}} \gtrsim 10^{-5}$, are large enough to survive the supernovæ explosions, which create and spread heavy elements. The second-generation stars and the planets form in these halos.

**The smallest halo mass**

The halo mass function in eq. (2.19) describes a distribution that extends, in principle, to arbitrarily small halo masses $\mathcal{M}$. However, because of the microscopic properties of the individual DM particles, which cannot be grasped by the continuum fluid approximation considered here, and which cannot be resolved by numerical simulations either, DM halos exist only above the *minimal viable halo mass* $\mathcal{M}_{\text{min}}$. The value of $\mathcal{M}_{\text{min}}$ depends on the speed of individual DM particles, rather than on their collective motion. Its computation requires concepts and assumptions discussed in chapter 4. The key points, however, can be sketched already now, working within two main assumptions. The first working assumption is that DM is made of particles that once were in thermal equilibrium with the SM components of the Universe, so that the DM velocity was fixed by $Mv_{\text{DM}}^2/2 = 3T$ in the non-relativistic limit. The second assumption is that the thermal equilibrium was lost when the universe cooled below some unknown temperature of kinetic decoupling, $T_{\text{kin dec}}$, which will be estimated in eq. (4.14) in terms of the DM particle properties. After kinetic decoupling, the DM velocity just red-shifts due to the expansion of the Universe, $v_{\text{DM}} \propto 1/a$.

Within this context, one can track the main physical processes at play, and compute $\mathcal{M}_{\text{min}}$, as follows. The sub-halos that are too small are washed out by the motion of individual DM particles in three different ways [39]:

1. At high temperatures, $T \gtrsim T_{\text{kin dec}}$, DM still interacts kinetically with the SM particles, and thereby undergoes diffusive Brownian motion with DM temperature equal to the temperature of the ordinary matter.

2. More importantly, after kinetic decoupling, $T \lesssim T_{\text{kin dec}}$, DM free-streams over a length $\lambda_{\text{fs}}$, which then defines the typical size of the minimal sub-halo and its mass at time $t_0$

$$\mathcal{M}_{\text{min}} \simeq \frac{4\pi}{3}\, \rho_{\text{DM}}(t_0)\, \lambda_{\text{fs}}^3, \qquad \lambda_{\text{fs}} = \int_{t_{\text{kin dec}}}^{t_0} dt'\, v_{\text{DM}}(t')\frac{a(t_0)}{a(t_{\text{kin dec}})} \sim \frac{M_{\text{Pl}}}{T_0\sqrt{MT_{\text{kin dec}}}}. \qquad (2.21)$$

3. The third mechanisms that suppresses small halos involves the acoustic oscillations produced in the DM fluid when it is still coupled to normal matter, analogously to the baryonic acoustic oscillations observed in the matter power spectrum.

Whether the last mechanism is more important or the previous two are, depends on the specific DM model. The free-streaming effect is often dominant for weak-scale DM, so we focus on it. Inserting in eq. (2.21) the kinetic decoupling temperature $T_{\text{kin dec}}$ as estimated in eq. (4.14) (the estimate assumes that the DM abundance arises from thermal freeze-out) gives

$$\mathcal{M}_{\text{min}} \simeq \frac{\Omega_{\text{DM}} H_0^2 M_{\text{Pl}}^{43/8} g_{\text{DM}}'^{3/2}}{2T_0^3 M^{15/8} M'^{3/2}} \sim 10^{-8} M_\odot \qquad \text{for} \qquad M \sim M' \sim 100\,\text{GeV} \quad \text{and} \quad g_{\text{DM}}' \sim \mathcal{O}(1). \qquad (2.22)$$

Note that $\mathcal{M}_{\text{min}}$ depends on the unknown particle physics: the DM mass $M$ and its coupling to matter $g_{\text{DM}}'$, mediated by some particle of mass $M'$. Qualitatively, a large DM mass $M$ implies a low typical speed of DM particles and hence small minimal halos $\mathcal{M}_{\text{min}}$; conversely, a large value of the combination $g_{\text{DM}}'/M'$ implies a tight interaction with normal matter, hence a late kinetic decoupling, a more effective suppression of small halos and therefore larger minimal halos $\mathcal{M}_{\text{min}}$ (see, e.g., Bringmann et al. (2016) in [63]). The spread in the possible values of the unknown particle physics parameters, and their prominence in determining $\mathcal{M}_{\text{min}}$, means that in practice $\mathcal{M}_{\text{min}}$ is very uncertain.

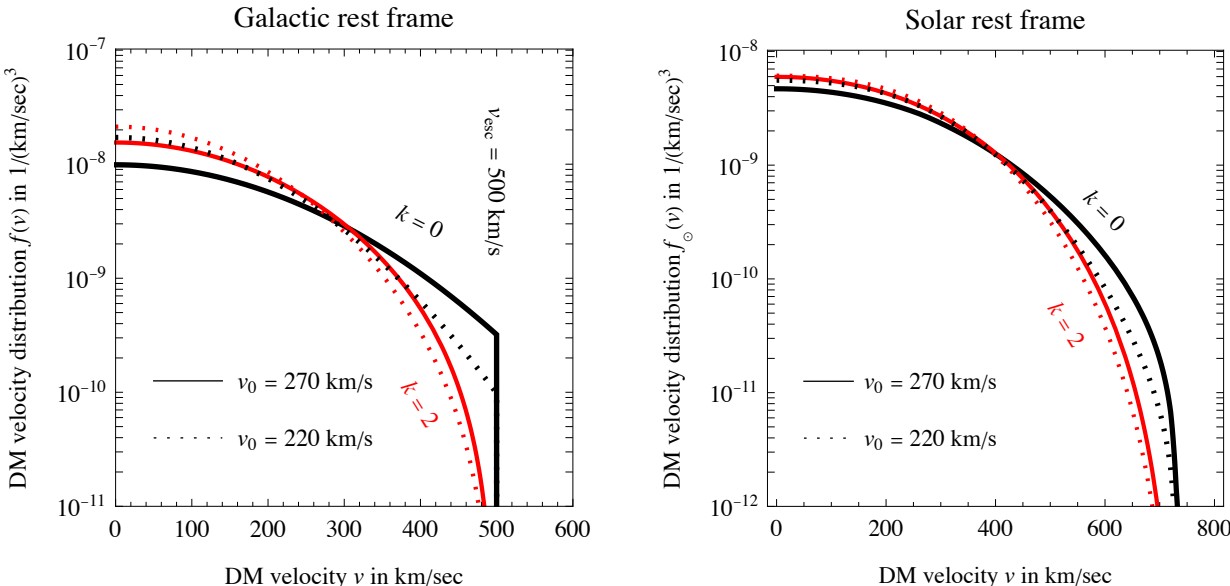

Figure 2.4: ***DM velocity distributions****: the Maxwell-Boltzmann distribution with a sharp cutoff at* $v = v_{esc} = 500 \, km/s$ *(thick black curve, $k = 0$), and the distribution motivated by the N-body simulations, eq. (2.26), with a smooth cutoff computed for $k = 2$ (red). Two different values of $v_0$ are shown:* $220 \, km/s$ *(dotted) and* $270 \, km/s$ *(solid). Left (right) panel shows the DM velocity distribution in the galactic (solar) rest frame (note the different scales on the horizontal and vertical axes in the two panels).*

## 2.3    Dark Matter velocity distribution

Next, we turn our attention to the DM velocity distribution $f(\boldsymbol{x}, \boldsymbol{v})$, where we will focus mostly on the Milky Way. The treatment can be easily extended to other galaxies, most notably to the dwarf galaxies, by replacing appropriately the parameters. Initially, we will neglect the dependance of DM velocity distribution on the position $\boldsymbol{x}$ (this is relaxed at the end of this section). Furthermore, the DM velocity distribution in the galactic rest frame is almost always assumed to be isotropic, so that $f(v)$ is a function of $v = |\boldsymbol{v}|$ only, and we often speak of a DM *speed* distribution.

As discussed in section 2.1.2, the Eddington formula relates a spherical density $\rho(r)$ to a spherical velocity distribution. This was used to show in eq. (2.8) that a Gaussian corresponds to an isothermal sphere $\rho \propto 1/r^2$. The procedure can now be inverted to find a velocity distribution corresponding to the more realistic profiles discussed in section 2.2.1. For example, a power-law $\rho \propto 1/r^\gamma$ (appropriate at small $r$) corresponds to $f(v) \propto v^{(\gamma-6)/(2-\gamma)}$. This provides an interesting approximation [36] despite that the assumptions behind the Eddington formula are not satisfied: the distributions are not strictly time-independent, and baryonic matter forms non spherical disks.

This discussion suggests that the DM velocity distribution cannot exactly follow a Gaussian Maxwell-Boltzmann distribution. In particular, DM particles that happen to acquire a speed larger than the galactic escape velocity, $v_{esc}$, tend to evaporate away. To account for this, the DM speed distribution in the galactic rest frame, $f(v)$, is often assumed to be an isotropic Maxwell-Boltzmann distribution that is sharply cut off at the finite escape speed

$$f(v) = N \, e^{-v^2/v_0^2} \, \Theta(v_{esc} - v) \, . \tag{2.23}$$

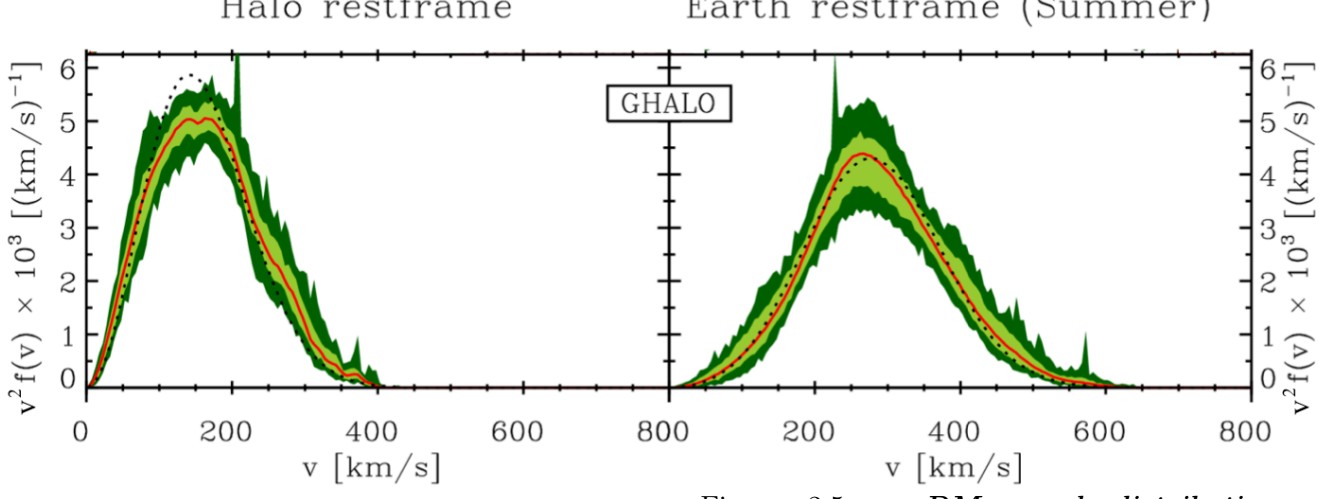

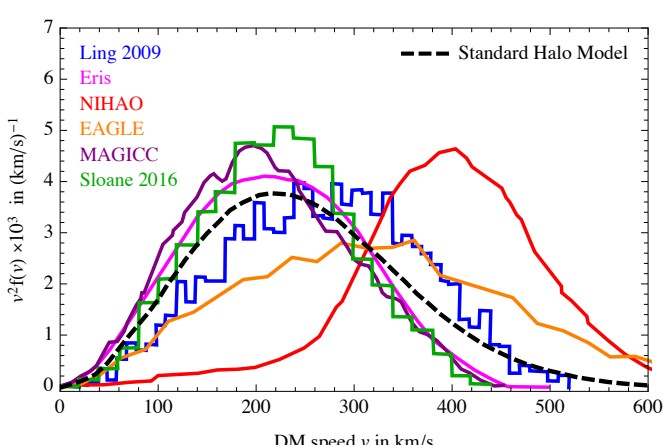

Figure 2.5: **DM speed distributions from sample numerical simulations.** *Top (adapted from Kuhlen et al. (2010) [64]): distribution in the galactic rest frame (left) and in the Earth's rest frame (right) as obtained from GHALO N-body simulation. The solid red line is the average distribution, with the light (dark) green shaded regions giving the 68% scatter (envelope). The dotted line is the best-fitting Maxwell-Boltzmann distribution. Left (adapted from Bozorgnia and Bertone (2017) [64]): distribution in the galactic rest frame for a sample of hydrodynamical simulations. The dashed black line corresponds to the Maxwell-Boltzmann distribution with the peak speed of 220 km/s.*

The normalization constant is fixed such that $\int d^3v\, f(v) = 1$, giving $N = \pi^{-3/2} v_0^{-3}$ in the limit $v_{\text{esc}} \gg v_0$ (here $d^3v = 4\pi v^2\, dv$). The parameters $v_{\text{esc}}$ and $v_0$ are estimated as follows:

○ The virial theorem $\langle K \rangle = -\frac{1}{2}\langle V \rangle$ applied to the isothermal sphere implies $2\sigma^2 = \langle v^2 \rangle = v_{\text{circ}}^2$ (see page 44) where $v_{\text{circ}}$ is the local circular velocity, measured to be $v_{\text{circ}} \approx (220 \pm 10)\,\text{km/s}$ [65] from motions of the stars. Since $\langle v^2 \rangle = 3v_0^2/2$ this suggests $v_0 = \sqrt{2/3}\, v_{\text{circ}}$. However, as discussed on page 44, the approximation of an isothermal sphere for DM distribution is not entirely realistic. In particular, it implies an infinite escape velocity, since the enclosed mass $\mathcal{M}(r)$ diverges for $r \to \infty$.

○ Observations of stellar motions show that the local[15] escape velocity from the Milky Way is [66]

$$\boxed{v_{\text{esc}} \approx (544 \pm 35)\,\text{km/s}}\,. \tag{2.24}$$

Measured speeds of old stars, which are expected to have a speed distribution similar to DM [67],

---
[15]That is, near the location of the solar system.

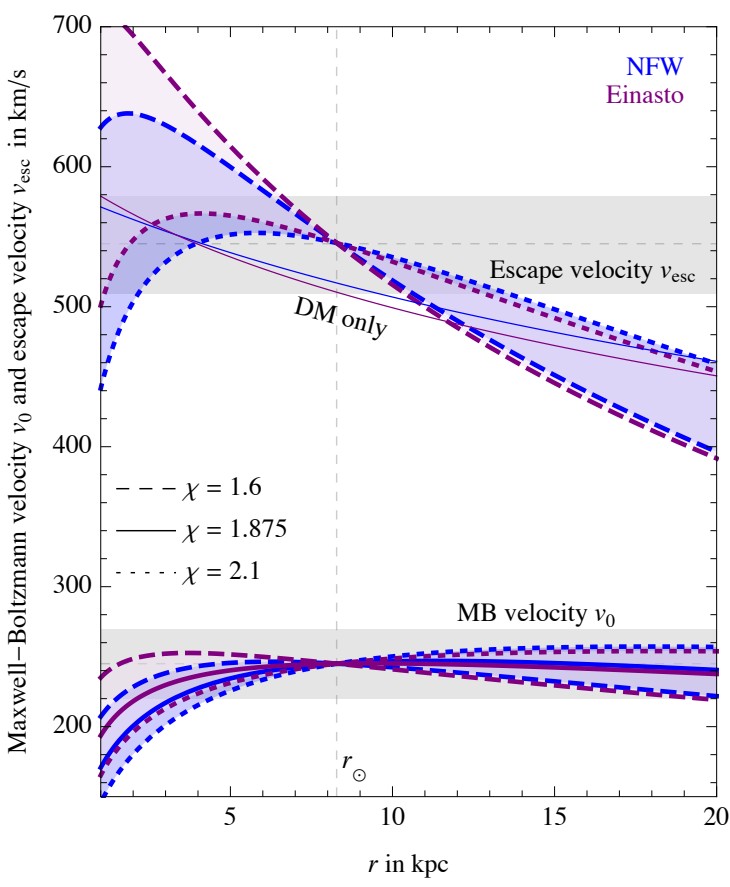

Figure 2.6: ***Illustration of the depen-dence of DM speeds on the position in the Galaxy:*** *the Maxwell-Boltzmann parameter $v_0$ in the bottom, and the escape velocity $v_{esc}$ in the top of the plot, accord-ing to the relations (2.27) and (2.28). The blue (purple) lines are for NFW (Einasto) DM profiles. The solid lines are for Bertschinger's $\chi = 1.875$, and dashed (dotted) lines for $\chi = 1.6(2.1)$. The nor-malizations are such that at the solar po-sition $r_\odot$ the $v_0$ and $v_{esc}$ coincide with the central values in eq.s (2.25) and (2.24), re-spectively, while the grey horizontal bands illustrate the plausible uncertainty ranges on these values. The thin solid lines in the upper part of the plot denote $v_{esc}(r)$ for a galaxy containing only DM.*

suggest

$$220\,\mathrm{km/s} < v_0 < 270\,\mathrm{km/s}\,. \tag{2.25}$$

$N$-body simulations indicate a more complex picture [64].

○ $N$-body simulations seem to suggest a distribution with bumps and dips, deviating significantly from a simple Maxwell-Boltzmann (see fig. 2.5). Furthermore, the distribution features a smoother cut-off at $v < v_{esc}$ and can be parameterised as (Lisanti et al. (2011), Kuhlen et al. (2010) [64])

$$f(v) = N_k \left[\exp\left(\frac{v_{esc}^2 - v^2}{kv_0^2}\right) - 1\right]^k \Theta(v_{esc} - v)\,, \tag{2.26}$$

with $1.5 < k < 3.5$. The Maxwell-Boltzmann distribution of eq. (2.23) is obtained in the limit $k \to 0$. These velocity distributions are plotted in fig. 2.4a.

○ More recent hydrodynamical simulations, which include baryons, seem to suggest instead a speed distribution closer to the standard Maxwell-Boltzmann, albeit with a $v_0$ that is by a few tens of km/s larger than the one in eq. (2.25). The spread among different simulations, as well as among different realizations of a single simulation is so large though, that a more precise determination of $v_0$ does not seem to be possible. Hydrodynamical simulations also seem to find a more isotropic distribution than the DM-only ones.

The average DM speed varies with the position $r$ in the Galaxy [68], see figure 2.6. Numerical

simulations suggest a simple power law scaling

$$v_0^3(r) \propto r^\chi \rho(r), \tag{2.27}$$

with the exponent $\chi \approx 1.9 - 2.1$ in the DM-only simulations, and $\chi \approx 1.60 - 1.67$ in the simulations that include baryons. In fact, already the pioneering work Bertschinger (1985) [68] predicted such a relation, with $\chi \approx 1.875$, based on spherically symmetric collapse.[16] The size of positional variation in $v_0$ depends on the assumed values of $\chi$ and the DM density profile $\rho(r)$, see eq. (2.27). In practice, however, $v_0$ does not change much, at least for $r$ within a few kpc of the location of the Sun, $r_\odot \simeq 8.3$ kpc. For the $\chi$ values obtained from the DM-only simulations, DM particles are on average slower in the center of the Galaxy, and faster on the outskirts, for $r$ beyond the position of the solar system. For the $\chi$ values that follow from the simulations that include baryons, on the other hand, the opposite tendency is found. The relation (2.27) holds over 2.5 orders of magnitude in $r$, from roughly $r \approx 0.1$ kpc (the resolution limit of the $N$-body simulations) up to at least $r \approx 20$ kpc.

The escape velocity $v_{\rm esc}$ is also clearly position depend. If one were to take just the DM into account, the escape velocity would have been straightforwardly predicted from its gravitational potential, which is determined by the DM mass enclosed within radius $r$. In the inner part of the Galaxy, however, the baryons constitute a significant fraction of matter content, which then makes the determination of $v_{\rm esc}(r)$ more involved. The following empirical parametrization is sometimes used (see Cirelli and Cline (2010) in [68]):

$$v_{\rm esc}^2(r) = 2v_0^2(r)\Big[2.29 + \ln(10\,{\rm kpc}/r)\Big]. \tag{2.28}$$

Using eq. (2.27) for $v_0(r)$, one can find that the escape velocity is typically highest in the inner galaxy, as expected, although the detailed behavior depends on the values of $\chi$ and $\rho$.

Extra possible complications, such as DM streams, will be discussed in section 2.4.5. Note that, while DM velocity distribution is rather uncertain, it is robust that the bulk of DM has $v \sim 10^{-3}$, and for many predictions this is good enough and the details are not so important. In other cases, these details matter and uncertainties on predictions can be very large. Among the many uncertainties, it is worth mentioning that extra DM particles with large speeds around $v_{\rm esc}$ could be present, if the Milky Way recently accreted extra DM, or even with speeds well above $v_{\rm esc}$, if DM is being boosted by interactions with cosmic rays (see section 5.5.4).

## 2.3.1 DM velocity with respect to the Sun

The previous subsection mostly dealt with DM velocity (or speed) distribution in the rest frame of the Galaxy.[17] Of more immediate phenomenological interest, however, is the DM velocity distribution in the solar local frame, $f_\odot(\boldsymbol{v})$. The $f_\odot(\boldsymbol{v})$ distribution, for instance, enters the estimates of DM capture by the Sun, and the subsequent neutrino flux (see section 6.9.1). The $f_\odot(\boldsymbol{v})$ is straightforwardly obtained from the velocity distribution in the galactic frame, $f(\boldsymbol{v})$, by performing a Galilean transformation $f_\odot(\boldsymbol{v}) = f(\boldsymbol{v} + \boldsymbol{v}_\odot)$ where

$$\boldsymbol{v}_\odot = (0, 220, 0)\,{\rm km/s} + (10, 13, 7)\,{\rm km/s}, \tag{2.29}$$

is the velocity of the Sun, here written as the sum of the local Galactic rotational velocity, and the Sun's peculiar velocity. We used Galactic coordinates, where $\hat{\boldsymbol{x}}$ points from the Sun toward the Galactic center, $\hat{\boldsymbol{y}}$ in the direction of disk rotation, and $\hat{\boldsymbol{z}}$ toward the galactic north pole. The modulus of $\boldsymbol{v}_\odot$ is $v_\odot \approx 233$ km/ sec.

---

[16]The formalism in Bertschinger (1985) is somewhat different from the one adopted here; see Taylor and Navarro (2001) for the recast. The ratio $\rho/v_0^3$ is called the (pseudo-)phase-space density.

[17]Our galaxy and the DM that it contains move at about 600 km/s with respect to the CMB rest frame. This movement, however, does not affect our relative velocity with respect to DM in the Galaxy and is therefore of little interest.

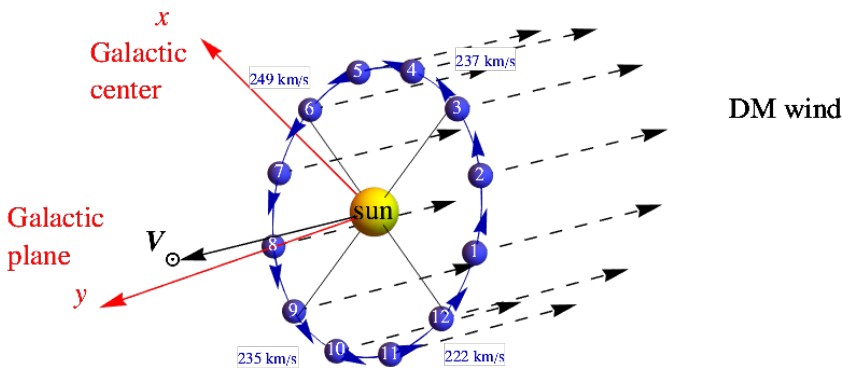

Figure 2.7: *The solar system and the Earth (blue spheres at indicated months) are moving through the stationary DM halo, which results in the effective '**DM wind**' in the Earth's frame. The DM wind is at the maximal speed, around $v_\oplus \approx 249$ km/s, at the beginning of June, when the Earth is moving fastest in the direction of the galactic disk's rotation, and at the minimal speed at the beginning of December, when the Earth is moving fastest in the opposite direction to the galactic disk's rotation.*

Below, we will also need the spherical angular average of $f_\odot$, which is given by

$$f_\odot(v) = \frac{1}{2} \int_{-1}^{+1} d\cos\theta \; f\left(\sqrt{v^2 + v_\odot^2 + 2vv_\odot \cos\theta}\right). \qquad (2.30)$$

The numerical results for several examples of $f_\odot(v)$ are plotted in fig. 2.4b.

## 2.3.2 DM velocity with respect to the Earth

The DM velocity distribution in the Earth's rest frame, $f_\oplus(v, t)$, is obtained in terms of the velocity distribution in the galactic frame, $f(\boldsymbol{v})$, through

$$f_\oplus(\boldsymbol{v}, t) = f(\boldsymbol{v} + \boldsymbol{v}_\oplus(t)). \qquad (2.31)$$

Here $\boldsymbol{v}_\oplus(t)$ is the relative motion of the Earth with respect to the galactic frame. It is given by

$$\boldsymbol{v}_\oplus(t) = \boldsymbol{v}_\odot + V_\oplus \left[\hat{\boldsymbol{\varepsilon}}_1 \cos\omega(t - t_1) + \hat{\boldsymbol{\varepsilon}}_2 \sin\omega(t - t_1)\right], \qquad (2.32)$$

where $\boldsymbol{v}_\odot$ is the velocity of the Sun, $V_\oplus = 29.8$ km/ sec is the Earth's orbital speed, $\omega = 2\pi/$ yr, $t_1 = 0.218$ yr = March 21 is the time of the spring equinox, while $\hat{\boldsymbol{\varepsilon}}_1$ and $\hat{\boldsymbol{\varepsilon}}_2$ are the orthogonal unit vectors tangent to Earth's orbit at spring equinox and summer solstice (at $t_1 +$ yr/4), respectively. In Galactic coordinates (defined in section 2.3.1, eq. (2.29)) one has

$$\hat{\boldsymbol{\varepsilon}}_1 = (\;\; 0.9931, 0.1170, -0.0103), \qquad \hat{\boldsymbol{\varepsilon}}_2 = (-0.0670, 0.4927, -0.8676). \qquad (2.33)$$

Consequently, the modulus of the Earth's velocity is

$$v_\oplus(t) = \sqrt{v_\odot^2 + V_\oplus^2 + 2bv_\odot V_\oplus \cos[\omega(t - t_c)]} \simeq v_\odot + \frac{V_\oplus^2}{2v_\odot} + bV_\oplus \cos[\omega(t - t_c)], \qquad (2.34)$$

where

$$b = \frac{\sqrt{(\hat{\boldsymbol{\varepsilon}}_1 \cdot \boldsymbol{v}_\odot)^2 + (\hat{\boldsymbol{\varepsilon}}_2 \cdot \boldsymbol{v}_\odot)^2}}{v_\odot}, \approx 0.490 \qquad (2.35)$$

is the sine of the angle between $\boldsymbol{v}_\odot$ and the normal to the orbital plane of the Earth, and

$$t_c = t_1 + \frac{1}{\omega} \arctan \frac{\hat{\boldsymbol{\varepsilon}}_2 \cdot \boldsymbol{v}_\odot}{\hat{\boldsymbol{\varepsilon}}_1 \cdot \boldsymbol{v}_\odot} \approx 0.415 \, \mathrm{yr} = \text{June 2}, \tag{2.36}$$

is the time at which $v_\oplus(t)$ is maximal, $v_\oplus(t_1) \approx 249 \, \mathrm{km/sec}$. The minimal Earth's velocity is $v_\oplus(t_1 + \mathrm{yr}/2) \approx 222 \, \mathrm{km/sec}$. The geometry of different contributions to $v_\oplus(t)$ is plotted in fig. 2.7. The time variation of $v_\oplus(t)$ has phenomenological consequences: it leads to the annual modulation of DM direct detection signals, as discussed in section 5.1.7.

## 2.4   Beyond the dark spherical (and isotropic) cow limit

To a first approximation the present day DM halos, at least the galactic ones, are often assumed by particle physicists to be spherically symmetric, static (therefore also in a steady state), homogeneous, and filled with particles obeying an isotropic velocity distribution. However, as we have encountered repeatedly above, this is just an approximation. A more detailed analysis shows that significant deviations from this picture can occur. We already discussed one important example: the expected presence of **subhalos**, i.e., clumps of DM (see section 2.1.3). Other deviations, discussed in this section, include departures from sphericity, from the assumption of static (and more generally, steady state) DM distributions, from the isotropy of DM velocities, etc. These deviations are not only relevant per se, since they are an essential stepping stone toward our understanding of the astrophysics of DM and how the microscopic properties of DM affect its distribution, but can also impact significantly the predicted signals in direct or indirect detection, see chapters 5 and 6.

### 2.4.1   Non-sphericity of DM halos

Numerical simulations consistently find DM halos with an ellipsoidal shape [69].[18] While the details can be complex (for one, it is not trivial to define the iso-density surfaces that identify the ellipsoid and its inner layers) and somewhat simulation-dependent, there seems to be consensus on a few general results. i) Galactic DM halos (and possibly also cluster halos) are triaxial, with a tendency towards prolate. Typical axes ratios are $b/a \sim 0.65$ and $c/b \sim 0.8$. The shape can vary mildly with the distance from the center: a halo can be thought of as a deformed-onion-like system with the inner shells typically more prolate, while the outer shells are progressively rounder; the shells are rather well aligned. ii) More massive halos are more triaxial. iii) The addition of baryons makes the halos rounder and more oblate. iv) The detailed dynamical mechanisms by which the triaxial shapes are achieved are still under study. They are probably related to the history of formation of the halos within a complex network of DM structures: the prolate direction might be a remnant of the incoming direction of DM particles from filaments, while the more isotropic accretion of late periods can result in rounder forms. In this sense, the shape of a halo can evolve as a function of time and redshift.

The results of observations via weak lensing of stacked galactic and cluster halos show some evidence for triaxiality too. For the specific case of the Milky Way, however, the situation is more confusing. One would normally expect for the Milky Way halo to be triaxial, with ratios of axes similar to those quoted above. However, the attempted measurements, using in particular the observations of stellar streams such as the Sagittarius stream or of stellar kinematics (notably from the GAIA mission), have resulted in indications of a halo that is anything from prolate to oblate to spherical.

---

[18]An ellipsoid is characterized by three semi-axes $a$, $b$ and $c$, with $a \geq b \geq c$ by convention. It is *prolate* (rugby ball shaped) if $a > b \approx c$, *oblate* (pancake shaped) if $a \approx b > c$, and *triaxial* otherwise. In terms of the triaxiality parameter $T \equiv (a^2 - b^2)/(a^2 - c^2)$ prolate means $T > 2/3$, oblate $T < 1/3$, and triaxial $1/3 < T < 2/3$.

## 2.4.2 Rotating DM halos

The issue of spinning DM halos, in particular those that surround the rotating disks of spiral galaxies, is intrinsically connected with the formation of the disks themselves, which is an active research field in astrophysics [34, 70]. The standard story is the one we have already alluded to in the beginning of this chapter, and goes roughly as follows. Initially, in the early stages of structure formation, both DM and baryonic gas were in the form of diffuse, roundish clouds. These clouds gained an initial non-vanishing angular momentum as a consequence of the torques applied by the tidal gravitational fields from neighbouring over-densities (a mechanism that goes under the name of Tidal Torque Theory). Subsequently, the baryonic gas, which is dissipational, i.e., it can radiate away electromagnetically part of its energy, started to cool and sink to the center. Barring significant inflows and outflows, it can be assumed that its angular momentum was conserved during the cooling. This means that the baryonic gas cannot collapse all the way to the center, but instead re-arranges itself into a rotationally supported spinning disk. DM particles, on the contrary, cannot dissipate energy (see page 16). DM clouds therefore shrunk in size,[19] which somewhat increased their rotational velocities, but the collapse came to a halt as soon as the systems virialized. As a result, the initial large and slowly spinning clouds morphed into the familiar configurations of a small rapidly rotating galactic disk surrounded by a much larger DM halo whose spin has not increased significantly. It is in this sense that the approximation of a static DM halo can be quite adequate.

The somewhat naïve picture above is essentially supported by the results of numerical simulations. The simulations also confirm that, as expected, the rotation axes of the disks and their host DM halos are roughly aligned, within at most 30° or so ('co-rotating DM halo'). The picture can be made more complex by several 'environmental' phenomena intervening at different stages of galaxy formation (cold inflows of baryonic gas depositing angular momentum in the forming disk, halos merging, satellite galaxy accretion,...) and/or by baryonic physics.

For a more quantitative treatment one can introduce the dimensionless *spin parameter* $\lambda$, which for any object such as a DM halo or a galactic disk is defined as $\lambda = j_{\rm sp}\, E_{\rm tot}^{1/2}/G\,\mathcal{M}_{\rm tot}^{3/2}$. Here, $j_{\rm sp}$ is the *specific* angular momentum, i.e., the total angular momentum of the system divided by its mass: $j_{\rm sp} = |\boldsymbol{J}|/\mathcal{M}_{\rm tot}$, where $\boldsymbol{J}$ is the total angular momentum,[20] $E_{\rm tot}$ is the absolute value of the total energy of the system (sum of kinetic and potential energy – note that this sum is negative), and $\mathcal{M}_{\rm tot}$ its total mass. The parameter $\lambda$ essentially measures the fraction of the total energy of the system that is stored in its spinning motion. It allows to compare consistently systems of different masses and sizes. For an isolated system undergoing gravitational dissipation-less evolution, such as the DM halo, $\lambda$ is conserved, since all the quantities entering its definition are conserved. Simulations consistently find $\lambda$ to be distributed around $\lambda_{\rm DM\ halo} \approx 0.04$, in agreement with theoretical expectations. In contrast, for a dissipational system such as the baryonic gas, $E_{\rm tot}$ increases since the binding energy increases when the cloud shrinks to form the disk. One finds $\lambda_{\rm disk} \approx 0.4$ at the end of the evolution. The result $\lambda_{\rm disk} \gg \lambda_{\rm DM\ halo} \neq 0$ expresses in formulæ that the rotation of the DM halo exists, but plays only a minor role.

The impact of bulk DM halo co-rotation (or even counter-rotation[21]) on direct detection signals has been phenomenologically investigated in a few works [70]. These found rather modest modifications of DM scattering rates, and thus also rather small changes in the derived bounds on the scattering cross sections (see chapter 5), from about 10% up to a factor $\mathcal{O}(2)$ in the extreme cases.

---

[19]Remember from section 2.1.2 that the virial theorem predicts the final halo radius to be $1/2$ of the 'turnaround radius'.

[20]Angular momentum is defined as $\boldsymbol{J} = \Sigma_i \boldsymbol{r_i} \times m_i \boldsymbol{v_i}$, where the sum runs over all the elements of the system, e.g., DM 'particles' or individual stars. The elements have masses $m_i$, are at positions $\boldsymbol{r}_i$ as measured from the system's center, and have velocities $\boldsymbol{v}_i$ in the center of mass system.

[21]A counter-rotating configuration can occur when massive inflows cause the baryonic disk to flip in the late stages of galaxy formation, something which is observed, albeit rarely, in simulations.

### 2.4.3 Dark disk?

Some numerical simulations find in their Milky Way-like simulated galaxies disks of Dark Matter that are aligned with the stellar disks [71]. Dark disks seem to be created by the disruption and dragging of DM satellites as well as simply through the gravitational pull of the baryonic disk. Their contribution to the local DM density varies between a few percent and 1.5 times the estimate for $\rho_\odot$ in (2.11), depending on the simulation. The disk is found in some cases to be dragged by the baryonic one and thus co-rotating, with a low lag speed of about 50 km/s. In other simulations, however, the dark disks are observed only very rarely, from which one would deduce that dark disks are unlikely to be relevant.

How likely it is for the dark disks to form also depends on the microscopic properties of DM. Some theoretical works speculated that a small fraction of Dark Matter (up to 5%) could consist of a dissipational and/or partially self-interacting substance that can therefore collapse and form a thin dark disk, parallel to the galactic disk [71]. This theoretical possibility was dubbed Double Disk Dark Matter (DDDM). If present, a dark disk would have an impact on DM direct and indirect detection. For instance, it would lead to enhanced rates for capture of DM particles inside the Sun and Earth (especially in the case of co-rotation, because low relative velocities make the capture easier), and therefore possibly to enhanced neutrino fluxes (see section 6.9). A more speculative effect on comet impacts and dinosaur extinction is mentioned in section 8.6.

Recent data from the GAIA mission indicate, however, that at most 1% of DM can be in the form of a thin dark disk [71].

### 2.4.4 Anisotropic DM velocity distribution

$N$-body simulations show that the orbits taken by dark matter particles are preferentially radial, resulting in an anisotropic velocity distribution [72]. Such a velocity anisotropy is not surprising for non-spherical halos, but it can also arise in halos exhibiting a spherically symmetric density profile.

More concretely, numerical simulations indicate that the radial velocity $v_r$ and the tangential velocity $v_t = \sqrt{v_\theta^2 + v_\phi^2}$ follow different profiles, approximated by non-Maxwellian distributions of the form

$$f(v_r, r) \approx N_r \exp\left[-\left(\frac{v_r^2}{v_{0r}^2(r)}\right)^{\alpha_r(r)}\right], \qquad f(v_t, r) \approx N_t v_t \exp\left[-\left(\frac{v_t^2}{v_{0t}^2(r)}\right)^{\alpha_t(r)}\right], \tag{2.37}$$

where $N_r$ and $N_t$ are the appropriate normalization factors determined as in eq. (2.23). Writing $v_{0r} = \beta_r(r)\, v_{\text{circ}}(r)$ and $v_{0t} = \beta_t(r)\, v_{\text{circ}}(r)$, the simulations suggest that at the position of the solar system [64]

$$\alpha_r \sim 1, \qquad \beta_r \sim 0.7, \qquad \alpha_t \sim 0.7, \qquad \beta_t \sim 0.4, \tag{2.38}$$

implying $v_{0t} \sim 120\,\text{km/sec}$, smaller than $v_{0r} \sim 200\,\text{km/s}$. All four parameters, $\alpha_r, \alpha_t, \beta_r, \beta_t$, become smaller at smaller $r$, in agreement with the discussion in section 2.3 for the case of DM only simulations. The velocity anisotropy parameter,

$$\beta_{\text{anis}} \equiv 1 - \frac{v_{0t}^2}{v_{0r}^2}, \tag{2.39}$$

equals $\beta_{\text{anis}} = 1$ in the limit where DM particles are all on radial orbits; $\beta_{\text{anis}} = -\infty$ for all tangential orbits; while the isotropic case corresponds to $\beta_{\text{anis}} = 0$. In general, $\beta_{\text{anis}}$ can vary as a function of $r$: the simulations find $\beta_{\text{anis}} \approx 0$ in the central regions of galactic halos, and $\beta_{\text{anis}} \approx 0.5$ in the outer regions.

### 2.4.5 DM streams

Galaxy formation is a continuous process: our galaxy is tidally stripping the smaller neighbouring galaxies and accreting mass to the expense of these smaller galaxies. Most probably, it is also tidally stripping and digesting its own DM subhalos close to the galactic center. These phenomena create streams of DM transpiercing the galaxy in various directions [73]. Some of these (such as the Sagittarius stream) were observationally identified thanks to the accompanying stellar flow. However, the numerical simulations predict the existence of many more DM streams.

DM streams are unlikely to contribute significantly to the local DM density: simulations find that they typically contain between 0.1% and 1% of the smooth average density of the underlying halo. However, they are directional and cold (i.e., characterized by low velocity dispersion). Therefore, they can modify significantly the DM velocity distribution from the one quoted in eq. (2.23) or in eq. (2.26) to $f_{\mathrm{halo+str}}(\boldsymbol{v}) = (1 - \rho_{\mathrm{str}}/\rho_{\odot})f(\boldsymbol{v}, v_0) + (\rho_{\mathrm{str}}/\rho_{\odot}) f_{\mathrm{str}}(\boldsymbol{v}_{\mathrm{str}})$. Here, $\rho_{\mathrm{str}}$ and $\boldsymbol{v}_{\mathrm{str}}$ are respectively the local DM density and velocity of the stream, and $f_{\mathrm{str}}$ is the functional form of the velocity distribution in the stream, which can be Maxwell-Boltzmann or not. Furthermore, one expects a small density $\sim 10^{-2-5}\,\mathrm{GeV/cm^3}$ of extra DM particles trapped in the Local Group and in the Virgo Supercluster to cross the Milky Way with higher velocities $\sim 600 - 1000\,\mathrm{km/s}$. Such an addition can be significant in the high-speed tail of the DM velocity distribution, possibly leading to important effects in DM direct detection.

It has also been argued that the DM galactic halos should exhibit **caustics**, i.e., accumulation surfaces of very high density, corresponding to the turnaround points for particles moving on gravitationally bound orbits (technically, the density is even divergent, in the limit of vanishing velocity dispersion) [74]. Caustics are expected to be a generic phenomenon in cold and collisionless fluids like DM, i.e., fluids with negligible velocity dispersion and negligible internal interactions. They would materialize as rings of enhanced density at specific radii in the inner and outer regions of the galaxy. Some evidence has been claimed for their existence in the Milky Way and in other galaxies, based on peculiar over-densities and regularly spaced bumps in the rotation curves [74].

### 2.4.6 DM around black holes

A higher DM density is expected to accumulate around black holes, which are known to exist with masses from few $M_{\odot}$ (stellar BH) to $10^{10}\,M_{\odot}$ (supermassive BH at the center of galaxies), including intermediate mass black holes (IMBH) with typical masses $10^2 - 10^5\,M_{\odot}$. This effect is usually approximated by defining the radius of gravitational influence $r_{\mathrm{in}}$ of the black hole as the radius within which the enclosed DM mass is twice the BH mass. Simulations suggest that a 'spiked' DM density profile starts inside $r_{\mathrm{sp}} \approx 0.2\,r_{\mathrm{in}}$ with power-law form [75]

$$\frac{\rho_{\mathrm{DM}}(r)}{\rho_{\mathrm{DM}}(r_{\mathrm{sp}})} \approx \begin{cases} (r_{\mathrm{sp}}/r)^{\gamma_{\mathrm{sp}}} & \text{at } r < r_{\mathrm{sp}}, \\ (r_{\mathrm{sp}}/r)^{\gamma} & \text{at } r > r_{\mathrm{sp}}, \end{cases} \tag{2.40}$$

where $\gamma_{\mathrm{sp}} \approx 1.5 - 2.5$, or perhaps $\gamma_{\mathrm{sp}} = (9 - 2\gamma)/(4 - \gamma)$ assuming adiabatic growth (Gondolo and Silk in [75]). Here $\gamma$ is the slope of the DM density profile ignoring the BH, for example $\gamma = 1$ for a BH located in the center of a galaxy with the NFW DM profile, as discussed in section 2.2.1. DM spikes can be softened or disrupted by stellar heating, galaxy mergers, etc.

If such DM spikes exist, they would make black holes good targets for DM indirect detection. Several groups have considered this possibility, in particular around the SMBH in the center of galaxies [75], around IMBH [76] and around PBH (see in this case the discussion in section 3.1.1). Chan and Lee (2023) in [77] claim indirect evidence for such DM spikes based on DM friction (section 2.5).

## 2.5   Dynamical friction

We end this chapter with a short discussion of the effect of the diffuse DM density on the motion of celestial bodies. The DM corpuscules exert a tiny but non-negligible gravitational force on bodies such as planets, stars and black holes. As realized by Chandrasekhar, this has a non-trivial consequence: a massive body moving with a certain speed with respect to ambient DM experiences a gravitational friction force known as *dynamical friction* [77] (this is in addition to the possible aerodynamical friction due to particle interactions).

The phenomenon can be qualitatively understood as follows: as the massive body plows through ambient matter, it attracts the ambient particles and they accumulate in its wake; this creates an over-density and thus a gravitational force that pulls the body back with respect to its direction of motion. Dynamical friction is sourced also by ordinary matter (e.g. diffuse interstellar gas), and it has the effect that stars tend to sink towards the galactic center, and planets toward their stars. Although ordinary matter is denser than dark matter in many environments, our focus in this discussion is on the DM-induced dynamical friction.

To provide a rough estimate of the effect, consider DM initially at rest at a distance $r$ from a body with mass $\mathcal{M}$ which is moving with speed $V$. DM falls onto the massive body in a free-fall time, $\Delta t \sim (r^3/G\mathcal{M})^{1/2}$. During the time interval $\Delta t$ the body attracts a mass $\Delta \mathcal{M} \sim r^3 \rho_{\rm DM} \sim G\mathcal{M}\rho\Delta t^2$, where $\rho_{\rm DM}$ is the DM density, and travels a distance $\Delta x = V\Delta t$. The consequent gravitational force is $F_{\rm DM} \sim -G\mathcal{M}\,\Delta\mathcal{M}/\Delta x^2 \sim -G^2\mathcal{M}^2\rho_{\rm DM}/V^2$.

More precisely, assuming that DM has a velocity distribution $f(v)$, and that it is made of bodies much lighter than $\mathcal{M}$, its dynamical friction is given by

$$\boxed{\boldsymbol{F}_{\rm DM} = -4\pi G^2\mathcal{M}^2\rho_{\rm DM}\frac{\boldsymbol{V}}{V^3}\ell\wp}\ , \tag{2.41}$$

where $\ell \sim 10$ is an infra-red enhancement due to the long-range nature of gravity and $\wp \le 1$ is the fraction of DM particles slower than $V$,

$$\wp = \int_0^V 4\pi v^2\, dv\, f(v) = {\rm erf}\frac{V}{v_0} - \frac{2V}{\sqrt{\pi}v_0}e^{-V^2/v_0^2}, \qquad \text{for } f(v) \propto e^{-v^2/v_0^2}, \text{ as in eq. (2.23).} \tag{2.42}$$

The consequent energy loss is

$$W_{\rm DM} = \boldsymbol{F}_{\rm DM}\cdot\boldsymbol{V} = -4\pi G^2\rho_{\rm DM}\mathcal{M}^2\wp\ell/V. \tag{2.43}$$

Dynamical friction is a purely gravitational effect that allows, in principle, to probe the very existence of DM (some authors challenge it claiming that its dynamical friction effect is not observed, see section 8.5.1), as well as its density and its velocity distribution.

Concerning probing the DM density, binary systems are particularly suitable: DM is denser and the energy loss due to its dynamical friction can compete with the energy loss due to the emission of gravitational waves, affecting the observable phase and/or the frequency spectrum (see section 8.5.4). Constraints on DM density have been derived from various sources: from binary pulsars ($\rho_{\rm DM} \lesssim 10^5\,{\rm GeV/\,cm^3}$), from binary solar-mass black holes (two anomalies are interpreted as a possible observation of DM with $\rho_{\rm DM} \sim 10^{11-13}\,{\rm GeV/\,cm^3}$), and from super-massive black holes [77].

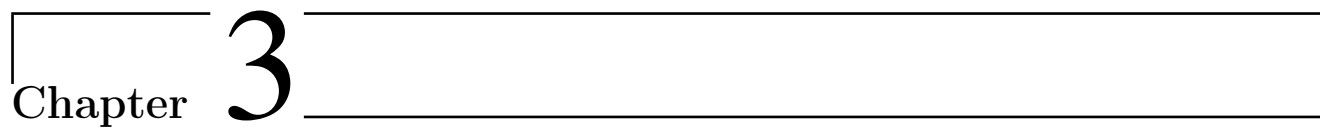

# Chapter 3

# What? Main paradigms regarding the nature of Dark Matter

In this chapter we review the main paradigmatic frameworks concerning the nature of Dark Matter. We start with some very general considerations (and speculations) on the available ranges of masses and sizes, and then we discuss the main basic properties of DM as a particle, as a macroscopic object or as an ultralight field.

Fig. 3.1 shows a catalogue of observed objects in the (mass, radius) plane $(M, R)$ (see [78, 79] for previous versions). The upper boundary of the allowed region, of triangular shape, is given by the cosmological horizon, $R \lesssim 1/H_0$. Its left boundary is given by quantum mechanics: at given $R$ *particle quanta* have the minimal mass $M \gtrsim 1/R$, in natural units $c = \hbar = 1$. Its right boundary is given by gravity: *black holes* have the maximal mass $M \lesssim M_{\rm Pl}^2 R$. The latter assumes that Einstein gravity holds up to the Planck scale, $M_{\rm Pl}$: this mass scale then represents the ultimate frontier of particle physics because any elementary particle with mass $M > M_{\rm Pl}$ would be a black hole, having a quantum wavelength $1/M$ smaller than its Schwarzschild radius, $R_{\rm Sch} = 2M/M_{\rm Pl}^2$.[1]

Normal matter forms solid objects around the blue dashed line in fig. 3.1, with roughly constant density $\rho \sim M/R^3 \sim \alpha^3 m_p m_e^3$ [81]. These objects range from close to the particle boundary (atoms made of electrons and protons) to molecules, viruses, cells, animals, asteroids, planets, stars up to the black hole boundary.

Nuclear matter forms objects around the red dot-dashed line, with constant density $\rho \sim m_p^4$, going from nuclei to neutron stars. In both cases, measuring the density of macroscopic objects and knowing quantum mechanics suggests fermionic particles with mass $\sim \rho^{1/4}$.

Observed objects with a significant component of Dark Matter, on the other hand, do not appear to exhibit a compact structure with some common density $\rho$. They rather seem to be gas-like structures with constant surface density $g \sim GM/R^2$ comparable to the Hubble acceleration $H_0$. Thereby their existence does not point[2] to a specific mass of DM particles $\sim \rho^{1/4}$.

---

[1] If gravity does not follow Einstein theory at high energies, the above statement may be revised. Experimentally, we have no evidence of how gravity behaves at energies above about an eV. Theoretically, Einstein theory does not give a renormalizable quantum theory. The black hole argument holds in string models of quantum gravity but is violated in models such as 4-derivative gravity [80].

[2] Ref. [79] takes a different perspective that dwarf galaxies, galaxies, clusters and superclusters do lie on the $M/R^3 = $ const. line. The authors then obtain $M \sim 1 - 100$ MeV as a plausible mass for the DM particle, by extrapolating towards the quantum boundary the equivalent of the dotted black line in fig. 3.1, in analogy with the lines for normal matter and nuclear matter. Since all DM dominated objects are of cosmological/galactic sizes this extrapolation is, however, over many orders of magnitude. See [79] for further discussion.

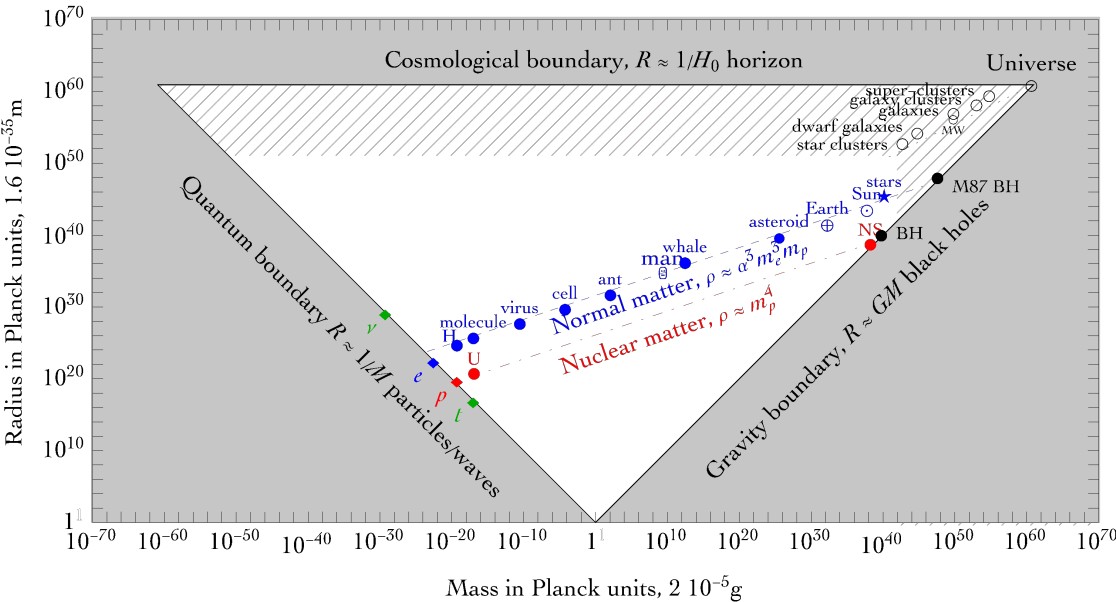

Figure 3.1: ***Catalogue of the Universe*** *in the plane* $(mass, radius) = (M, R)$. *Normal matter forms objects around the blue dashed line, i.e. with density* $\rho \sim M/R^3 \sim \alpha^3 m_p m_e^3$. *Nuclear matter forms objects around the red dot-dashed line, with density* $\rho \sim m_p^4$, *going from hadrons and nuclei to neutron stars, close to the black hole limit. DM is believed to be relevant for understanding the largest structures of the Universe, located around the black dotted line on the top right corner of the* $(M, R)$ *plane. They have constant surface gravity* $GM/R^2 \sim H_0$ *and thereby non-constant density* $\rho \propto 1/R$. *As discussed in this section, DM as an (elementary or composite) corpuscle cannot reside in the hatched region, because it would be larger (top bound) or heavier (right-hand side bound) than a small galaxy.*

If DM is a particle with mass $M$, it will be located somewhere around the quantum mechanics boundary, possibly up to $M_{\rm Pl}$ which, as said above, is the limit for a particle. Empirically, the DM particle have to be heavier than $10^{-21}$ eV $\approx 10^{-49} M_{\rm Pl}$. This lower limit is determined by the request that the de Broglie wavelength $R = 2\pi/Mv$ of a DM particle fits within the small gravitationally bound dwarf galaxies (which typically have kpc size, velocity $v \sim 10$ km/s, mass $\sim 5 \times 10^5 \, M_\odot$ where $M_\odot = 1.9984 \times 10^{30}$ kg is the solar mass) [82].

DM can be heavier than the Planck mass, if it takes the form of a composite object, possibly up to the gravitational boundary at which point black holes are the fundamental objects. In such a case, an upper bound is provided by the fact that the DM mass must be somewhat smaller than the mass of a typical small dwarf galaxy: since these have masses of the order of few $10^5 \, M_\odot$ (see section 2.2.3), one can conservatively impose $M \lesssim 10^4 \, M_\odot$, to make sure that a sufficient number of DM 'particles' inhabit the dwarfs. It is worth emphasizing that the above absolute lower and upper limits of allowable DM mass $M$ arise from rather robust and model-independent considerations since they use just the basic properties of dwarf galaxies, the smallest astrophysical structures known to host Dark Matter.

The 'in-principle-viable' range of DM masses therefore spans more than 90 orders of magnitude[3]

$$10^{-21} \, \text{eV} < M < 10^{37} \, \text{kg}. \tag{3.1}$$

---

[3]One can similarly discuss the possible range for the strength of the interaction between DM and ordinary matter, which spans about 20 orders of magnitude from the minimal gravitational interaction $g \sim M/M_{\rm Pl}$ up to strong-interaction-like couplings $g \sim 4\pi$. See Buckley and Peter [1] for the graphical representation of this range.

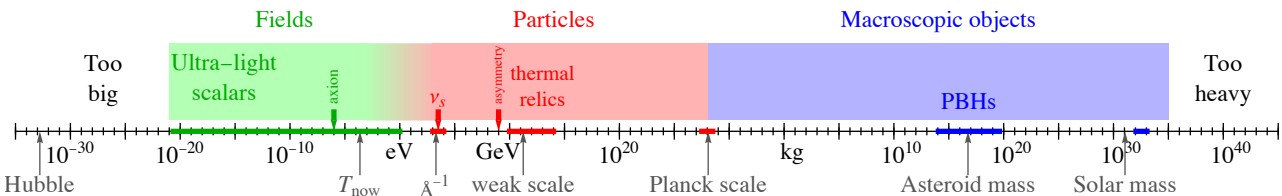

Figure 3.2: *Possible **range for the DM mass**, and some notable candidates. The edges of the shaded areas correspond to the lower and upper bounds in eq. (3.1).*

This huge range can be conceptually split in three main qualitatively different regions, illustrated in fig. 3.2: fields, particles and macroscopic objects. Fundamentally, particles and waves (fields) are the same objects, since Quantum Field Theory unifies them in a common description. From a practical point of view, however, descriptions using particles or waves are different enough to be useful in different regimes (see section 3.4). As a rule of thumb, Dark Matter behaves as a classical field if $M \ll$ eV, and as a particle if heavier than the inverse Bohr radius $M \gg \alpha m_e \sim$ keV. Dark Matter with de Broglie wavelength much smaller than atoms interacts with atoms individually, as a particle. In the opposite limit DM undergoes collective interactions with ordinary materials. The particle/wave transition similarly occurs in galaxies, where the observed DM density can be reproduced by particles lighter than about an eV only if many quanta occupy the same phase space volume, as we discuss shortly below. In this case DM can be described by a classical field. This is in complete analogy with electro-magnetism, where in the limit of many photons these are more simply described by classical electric and magnetic fields. Similarly, DM could be a massive boson that, in dense environments, is more simply described through a classical field.

In other words, from the outset we do not know whether DM physics belongs to astrophysics, particle physics or classical field theory. The three possibilities are described in more details in the following sections:

1. Section 3.1 and section 3.2 discuss DM as composite objects heavier than the Planck scale. Primordial black holes are one possible candidate.

2. Section 3.3 discusses DM as a new particle with mass $M$. In this case a plausible argument (see section 4.1) favors $M \sim$ TeV — the mass range currently explored by colliders and many other experiments — but the possible candidate masses span many orders of magnitude below and above TeV.

3. Section 3.4 discusses DM as waves of ultra-light bosons, a possibility that is now also very actively investigated.

## 3.1 DM as very massive macroscopic objects

DM could be made of Massive Astrophysical Compact Halo Objects (MACHOs[4]), i.e., ordinary astrophysical objects of macroscopic mass $M$, such as large planets, small dead stars or stray black holes [83]. These objects do not emit light and therefore fulfill the definition of dark matter. The MACHOs that are composed of baryonic matter *and* were created in the late Universe, like all the other astrophysical

---

[4]The name was coined in the early '90s (see K. Griest (1991) in [83]), in witty opposition to WIMPs, cf. section 9.3.3.

objects (the most natural expectation), require a large baryonic abundance, which contradicts the bounds from BBN and CMB discussed in section 1.3. DM made purely of *baryonic* MACHOs is thus ruled out. Nevertheless, it is still illustrative to consider their role as DM candidates, even though it is now mainly for historical reasons.

One way of identifying the presence of MACHOs in the Milky Way halo is via **gravitational microlensing**. When a compact object with mass $M$ happens to cross the line of sight between the observer and a background star located at distance $L$,[5] the light of such a star is lensed and its flux towards the Earth temporarily increases (two other related effects — creating a second image or modifying the apparent size of the star — are typically too small to be observable).

Since the time that MACHOs were first proposed in the '80s [83] a number of surveys such as Eros-1, -2, Macho, Ogle-I, -II, -III, -IV, tried to detect lensing signals by monitoring for several years millions of stars in the Magellanic clouds, which are known environments that are relatively nearby, just outside our own Galaxy [84]. More recently, similar analyses were performed with the Kepler data, using local stars, and with the Subaru HSC camera, using stars from the neighbouring galaxy Andromeda. The results by the Macho collaboration in the year 2000 generated significant excitement, when it initially reported that between 8% and 50% of the mass in the Milky Way halo could be due to massive objects with preferred mass of about half $M_\odot$. However, this possibility is excluded by other surveys which found only upper limits on $f$, the fraction of dark mass in the halo that is in the form of massive objects.[6] The recent reanalysis by Blaineau et al. (2022) [84] of archival data from the Eros-2 and Macho surveys has focused on long duration microlensing events and has allowed to derive new constraints at large mass $M \gtrsim 10\,M_\odot$. The recent results by Ogle (2024) and Ogle (2024b) [84] combined about 20 years of observations as well as high-cadence observations and derived stringent bounds in the wide range $10^{-9}M_\odot \lesssim M \lesssim 10^2 M_\odot$. Combining the results of different campaigns gives

$$f \lesssim 5\% \quad \text{for} \quad 10^{-11}\,M_\odot \lesssim M \lesssim 100\,M_\odot$$
$$\text{(exclusion by Eros, Ogle, Macho, Kepler, Subaru HSC).} \tag{3.2}$$

The detailed bounds are reported in fig. 3.3 (left). They hold under standard assumptions for the density and the distribution of the lensing objects, but could be weakened if, for instance, MACHOs are clustered (since this reduces the relative probability of them crossing a line of sight) or if their velocity dispersion is small. However, such effects are typically not sufficient to significantly weaken the bounds (see Green (2017) in [85]). Other surveys, some still ongoing or upcoming, include Moa, Super-Macho, the Vera Rubin Observatory (formerly Lsst), ... [84]. That there are possible excesses of microlensing events was suggested by Niikura et al. (2019) and by Meneghetti et al. (2020) in [84].

Besides bounds from microlensing, there are also other constraints on MACHOs [84], a selection of which are given in fig. 3.3 (right). The non-observation of lensing effects in gamma ray bursts (femtolensing of GRBs) was originally used to disfavor MACHOs in the mass range $5 \times 10^{-17} - 2.5 \times 10^{-14}\,M_\odot$. This conclusion has been disputed by Katz et al. (2018) and then Niikura, Takada, Yasuda et al. (2019) [84], who pointed out that for these small MACHO masses the wavelength of the incoming light is larger than the Schwarzschild radius of the lensing object. This means that the diffraction due to the wave properties of light needs be taken into account and the geometrical optics approximation used in the original GRB studies becomes invalid. For the same reason the Subaru HSC microlensing constraints cannot probe MACHOs with masses below $\sim 10^{-11}\,M_\odot$. For much higher masses, the absence of lensing in the direction of type-Ia supernovæ (SN-Ia) is claimed to exclude a large portion between $10^{-3} - 10^4\,M_\odot$, however, this claim is also disputed. At even higher MACHO masses, the absence of lensing for compact radio sources excludes the mass range $10^6 - 10^8\,M_\odot$.

---

[5]More precisely, lensing occurs when the compact object is within a distance $\sim \sqrt{GML}$ or less from the axis of the line of sight.

[6]An implicit assumption is that the fraction $f$ is the same in the whole Universe.

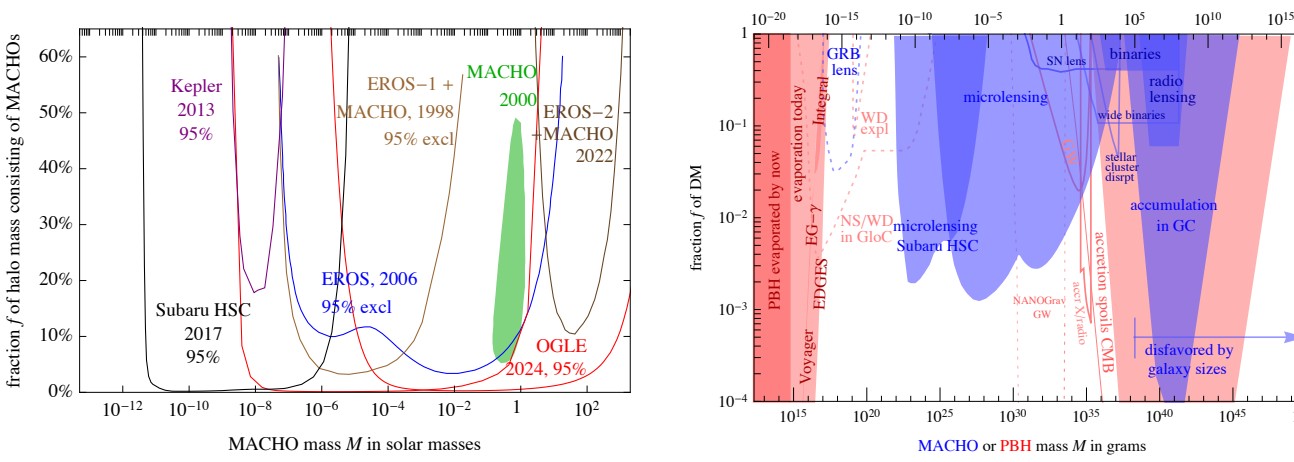

Figure 3.3: **MACHO searches. Left:** *Results of micro-lensing surveys (mostly towards the Magellanic Clouds): the bounds on the fraction f of the Milky Way halo's mass which can be due to MACHOs, as a function of the object's mass M, as well as the region identified by the MACHO collaboration in 2000 (green).* **Right:** *A collection of bounds on the fraction f of DM consisting of massive astrophysical objects. The blue bounds apply to any MACHO, including Primordial Black Holes (PBH). The red bounds apply only to PBHs. The most recent or the most debated bounds are shown as dashed lines or as open (unshaded) contours. For a compilation of current constraints see also* B. Kavanagh, PBHbounds.

Very massive MACHOs are subject to many other constraints due to possible dynamical effects. Requiring that the binary star systems observed in the Galaxy are not disrupted by encounters with MACHOs rules out a mass range $10^3 - 10^8 \, M_\odot$. The constraint extends to $10^{10} \, M_\odot$ (not shown in fig. 3.3), if the survival of globular clusters is taken into consideration as well. Recent results on wide halo binaries even lower the bound to $10 \, M_\odot$. In a similar spirit, requiring that the stellar clusters observed at the center of ultra-faint dwarf galaxies, in particular Eridanus II, are not disrupted by the passage of MACHOs can impose a bound that, in its most conservative version, rules out a portion above $100 \, M_\odot$ (recent simulations strengthen that to about $1 \, M_\odot$, see Koulen et al. (2024) [84]). Imposing that the amount of MACHOs being dragged (along with the stars) by dynamical friction into the galactic center region (see section 2.5) does not exceed the observed mass of the GC region itself, disfavors the large portion around $2 \times 10^4 - 10^{12} M_\odot$. This constraint is sensitive to the details of galactic dynamics, though. For instance, an in-falling MACHO could kick stars or other MACHOs out of the GC via gravitational slingshot effects, reducing the accumulation rate. Another bound excluding MACHOs with masses $10^4 - 10^{10} \, M_\odot$ (not shown in fig. 3.3) comes from requiring that DM behaves like a fluid, and not as a collection of discrete massive objects, in the formation of Large Scale Structures. Finally, the very heavy MACHOs in fig. 3.3 (right) are highly disfavored simply because a single DM 'particle' cannot be heavier than the smallest of the objects formed by DM particles, as discussed around eq. (3.1): the sizes of dwarf galaxies such as Segue 1 ($\sim 5 \times 10^5 \, M_\odot$) or even of the Milky Way ($\sim 5 \times 10^{11} \, M_\odot$) arguably impose robust upper bounds on MACHO masses, unless one is willing to allow tailor-made MACHOs for each galaxy... More stringent bounds could be derived by imposing that a sizable number of DM 'particles' must constitute the halos of these objects (to avoid granularities) or by detailed modelling of even smaller bodies (e.g., globular clusters).

In summary, the various observational and dynamical bounds only leave open the region at small MACHO masses ($M \lesssim 3 \times 10^{-12} M_\odot$). If the bounds from supernova lensing, the wide binaries and Eridanus II are discarded, and before the recent extension of the bounds using archival data from the EROS-2 and MACHO surveys, the region of MACHO masses around 10 to 400 $M_\odot$ was viable as well.

The latter possibility attracted significant attention in the wake of the gravitational wave detection by LIGO, as we mention in the following subsection.

However, let us recall: all of the above discussion is subject to the fact that the BBN and CMB constraints contradict the existence of baryonic bodies of astrophysical nature, of any mass, as an explanation for DM abundance, if these formed after the BBN.

### 3.1.1   Primordial black holes

Astrophysical objects that consist of baryonic matter, but have somehow been created *before* the BBN, are not subject to the cosmological constraints of section 1.3, since the material that they are made of gets subtracted from the baryonic budget very early on. This is the case with *primordial* black holes (PBHs) [86].[7]   PBH formation mechanisms will be discussed in section 4.6, and possible underlying theories in sections 4.6.1, 4.6.2 and 9.1.6. Here we discuss their viability as DM candidates.

Several **phenomenological constraints** apply to PBHs as candidates for DM [85], shown in fig. 3.3 (right). First of all, the bounds on MACHOs discussed in section 3.1 rely only on their gravitational pull and thus also apply to PBHs.

An additional condition is that the PBHs should not have evaporated by now. Quantum fluctuations of fields around any black hole lead to the emission of *Hawking radiation*, a thermal spectrum of particles at a temperature[8]

$$T_{\mathrm{BH}} = 1/(8\pi GM) . \tag{3.3}$$

The total radiated power is $W \approx g\, 4\pi R_{\mathrm{Sch}}^2 \sigma T_{\mathrm{BH}}^4 = g/(15360\,\pi\, G^2\, M^2)$ where $g$ is the number of degrees of freedom lighter than $T_{\mathrm{BH}}$, $\sigma = \pi^2/60$ is the Stefan-Boltzmann constant and $R_{\mathrm{Sch}} = 2GM$ the Schwarzschild radius. The BH loses mass at the same rate, $\dot{M} = -W$, so that within time

$$\tau_{\mathrm{BH}} \approx 5120\,\pi\, G^2 M^3/g, \tag{3.4}$$

the BH fully evaporates. Evaporation happens when the age of the universe is comparable to $\tau_{\mathrm{BH}}$, corresponding to the cosmological temperature $T_{\mathrm{ev}} \sim M_{\mathrm{Pl}}^{5/2}/M^{3/2}$. Requiring that the PBH's lifetime is longer than the age of the Universe implies $T_{\mathrm{ev}} \lesssim T_0$, where $T_0 \simeq 2.7\,\mathrm{K}$ is the current CMB temperature, so

$$M \gtrsim M_{\mathrm{Pl}}^{5/3}/T_0^{2/3} \sim 10^{-19}\, M_\odot. \tag{3.5}$$

---

[7]An interesting questions is whether one can distinguish observationally a Primordial BH from a standard BH created in a stellar collapse (e.g., via the signal of gravitational waves emitted in BH mergers). In principle, there are several possibilities: i) detection of a BH with a mass smaller than about $3M_\odot$ would imply that a PBH was observed, since astrophysics predicts that for such light objects the gravitational collapse is stopped by neutron degeneracy pressure and a neutron star is born (for a nonstandard exception see [87]); ii) detect a BH at very high redshift (say, $z > 8$), an epoch before the birth of first stars; iii) find evidence for a spatial distribution of BHs not correlated with the galactic disks.

[8]Although the phenomenon of BH Hawking radiation has never been confirmed observationally, there are strong theoretical and experimental arguments in its favor (for instance, the equivalent of a Hawking radiation was observed in fluido-dynamical systems that share some properties with the BHs). Let us however note that several recent works suggested that Hawking evaporation slows down or even stops as the BH's mass decreases, due to a quantum memory burden effect [88]. If this were indeed the case, evaporation constraints would no longer apply or at the very least would be relaxed over vast portions of the parameter space, which would thus be allowed again. Note also that the effect implies that two BH with the same mass would evaporate at different rates, if they have a different past history, contradicting the 'no-hair' principle. Other conditions in which Hawking radiation is argued to be slowed down or stopped are rotating (and/or electrically charged) BH, as well as BH in extra-dimensional spaces [89]. Finally, let us mention that, according to [90], the gravitational collapse of objects with higher central density might give rise to *primordial naked singularities*, that could behave as DM similarly to PBH, but without emitting Hawking radiation.

A detailed computation gives $M > 2.5 \times 10^{-19} \, M_\odot$.[9] PBHs with a mass slightly above this value would be in the process of evaporating at the present time, emitting particles with a temperature of around 80 MeV. The non observation of such Hawking radiation, e.g. in the extragalactic $\gamma$-ray flux or in the flux of low energy electrons and positrons measured by the VOYAGER-1 spacecraft outside of the solar system, imposes the more stringent bound $M \gtrsim 4 \times 10^{-17} \, M_\odot$. In addition, the emitted low energy $e^\pm$ would diffuse and annihilate. Requiring that the resulting contributions to the $X$-ray signal measured by INTEGRAL are not too large extends the bound by a tiny bit to $M \gtrsim 6 \times 10^{-17} \, M_\odot$.

Similarly, the evaporation of PBHs around the epoch of reionization would have heated up the intergalactic medium and weakened the 21cm absorption signal, claimed by the EDGES experiment. This provides a bound around $M \sim 10^{-16} \, M_\odot$ (denoted as EDGES in fig. 3.3 right) and possibly around 10 $M_\odot$ (the latter, not shown in fig. 3.3 right, should not be confused with the bounds imposed by the CMB due to accretion, discussed two paragraphs below). The heating of the intergalactic medium can also be constrained via the Lyman-$\alpha$ forest and by its effect on the CMB anisotropies as observed on Earth. These effects provide limits around $M \sim 10^{-16} \, M_\odot$ similar to the EDGES one, hence for simplicity we do not report them in the figure.

Constraints based on the survival rates of neutron stars (NS) of white dwarfs (WD) in Globular Clusters (GloC) apply to the PBH masses in the range from $10^{-18} \, M_\odot$ to $10^{-9} \, M_\odot$. Such PBHs could sink to the interior of NS/WD, either during the formation of the NS/WD or by subsequent capture, and quickly destroy them by eating them from the inside. The mere observation of existing neutron stars and white dwarfs then imposes the constraint on PHBs. However, such bounds rely on the assumption of a very large density of DM in Globular Clusters (of the order of $10^3$ GeV/cm$^3$ and higher), which is far from certain. For this reason the bound is indicated as an open dashed contour in fig. 3.3.

Using similar physics, the PBHs in the mass range between $5 \, 10^{-15} \, M_\odot$ and $10^{-13} \, M_\odot$ were claimed to be excluded based on the fact that PBHs piercing the bodies of white dwarf stars (WD) could heat the stellar matter and trigger the death of WDs in a form of supernova explosions at a rate higher than observed. However, more detailed numerical computations invalidated this bound.

Finally, PBHs, like any other BH, would accrete material. On the one hand, this could help explain why some dense structures are heavily dominated by DM: the PBHs may have swallowed a lot of the baryonic material early on, eventually penalizing the star formation, as argued by Clesse & García-Bellido (2018) [86]. On the other hand, the infalling gas would emit $X$-rays, which would ionize matter, and thus affect the observed CMB spectrum and anisotropies. This rules out a very large range of PBH masses around $10^2 - 10^{16} \, M_\odot$.[10] By refining the accretion model, Poulin et al. (2017) [85] improved the exclusion on the low mass end by about 2 orders of magnitude (open pink contour in fig. 3.3 right). The same $X$-ray (and radio) emissions from accretion should be visible if PBHs are present in our Galaxy. Their non-observation imposes a bound in the PBH mass window from a few $M_\odot$ to $100 \, M_\odot$ and possibly beyond. Importantly, all these bounds are subject to the uncertainties related to the modelling of the accretion process, which in itself is an active area of research in astrophysics, and by no means a settled subject, especially in the conditions encountered for these BH masses ($1 - 100 \, M_\odot$), gas densities ($\gtrsim 200$ particles/cm$^3$ at CMB formation) and relative PBH/gas speeds (super- or sub-sonic depending on the

---

[9] A possible loophole to this argument occurs if PBHs evaporate via Hawking radiation, but then leave behind *stable Planck-mass relics*, which do not evaporate further. Such remnants could constitute the DM [91] (as long as the last stages of the evaporation do not provide them with a momentum kick that would make them non-cold, as argued in some models). With an estimated cross section of $1/M_{\rm Pl}^2 \sim 10^{-66} \, {\rm cm}^2$, they would be essentially inert and undetectable (other than by their gravitational effects, see section 5.5.9). It has also been argued that the relics could carry an electric charge, if evaporation halts abruptly in a non-zero-charge configuration. This possibility is strongly constrained, see section 3.3.2. Alternatively, doubts related to unitarity suggest that after some time the semiclassical approximation that leads to Hawking radiation could fail, and maybe the BH lifetime is much longer than $\tau_{\rm BH}$.

[10] The early computation in Ricotti et al. (2007) [85], which found more constraining bounds down to $10^{-1} \, M_\odot$, was later revisited and the bound relaxed.

models).

Incidentally, the $10-100\,M_\odot$ mass interval attracted a lot of attention recently: the Ligo gravitational wave events [92] have been advocated by some groups to be due to mergers of PBHs [93]. However, for PBHs with such masses to be viable DM candidates the bounds from long duration lensing events in Eros-2 and Macho, from accretion, from SN lensing, from wide binaries and possibly from CMB distortions and stellar cluster disruptions in Eridanus II would all have to be dismissed. A few works [93] instead used the same Ligo observations, along with the estimates of PBH merger rates, to derive the exclusion contours shown in fig. 3.3 with the GW label (note, though, that the existence of such bounds has been questioned by others). An even wider range of masses $(3 \times 10^{-4} M_\odot < M < 2 M_\odot)$ was claimed to be strongly excluded by the fact that NANOGrav has not yet detected gravitational waves, even though these should in principle have been produced at the same time as the PBHs were created. The exclusion relies on extreme assumptions about the PBH production mechanism, has been questioned in the literature, and we thus show it as a dotted contour in fig. 3.3.

In summary, being very conservative when taking into account the constraints for different ranges of PBH masses, we find that at the time of writing the PBHs could constitute the whole of Dark Matter $(f = 1)$ for

$$10^{-16}\,M_\odot \lesssim M \lesssim 3 \times 10^{-12}\,M_\odot. \tag{3.6}$$

Black holes at the low end of this range would have radii smaller than the size of an atom and mass comparable to a small asteroid or Mount Everest. On average, roughly one such PBH would be expected to be present in the volume of the solar system. Also note that many of the bounds are still under active discussion, as mentioned above, so that the situation could well change.

It is important to stress that the above results apply under a couple of important simplifying assumptions. The first one is that all the PBHs have the same mass, i.e., a 'monochromatic mass function'. In the more general, and probably more realistic, case in which the PBHs masses are distributed according to an **extended mass function**, each mass could in principle lie below the various constraints and they could sum up to the needed total amount. The analysis is however complex since the monochromatic bounds cannot be applied straightforwardly, the constraints need to be re-evaluated at all the masses included in the specific mass function. Current results seem to indicate that PBHs with some extended mass functions (e.g. lognormal) might still account for the whole of DM in some restricted windows [94]. As discussed in section 4.6.1, many mechanisms can give rise to PBHs masses that span about an order of magnitude.

The second simplifying assumption is that PBHs are homogeneously distributed in space. If they instead wander around in the galactic halo grouped in **clusters**, as their formation history might suggest, the bounds might be affected. On the one hand, the lensing limits are expected to be relaxed, since the cluster tends to behave as a single lensing body of larger mass than the constituent PBHs, therefore falling 'to the right' of the region in mass probed by the lensing surveys. CMB limits could also be loosened, since the high proper velocity of the PBHs within the cluster implies a less efficient accretion. On the other hand, the tight packing inside clusters is expected to favour merging events, therefore enhancing the emitted signals in gravitational waves and strengthening the corresponding constraints. The extent to which clustering occurs and its detailed impact on the different bounds are subjects of active investigations [95].

One can also imagine more complicated scenarios. For instance, some works explored a hybrid hypothesis, where the total DM abundance would be a mix of PBHs and particle DM (possibly produced by the PBHs themselves via evaporation), in varying proportions [96]. However, quite often the two are mutually exclusive. For instance, in the hybrid scenario where DM are WIMPs, see section 6, the WIMPs would accumulate around PBHs and tend to produce a DM annihilation signal (see section 2.4.6) incompatible with observations. The DM abundance must in that case be either predominantly due to PBHs or predominantly due to WIMPs, see e.g. Carr, Kuhnel & Visinelli (2020) in [96].

Future prospects to further explore the PBH parameter space include the following ideas [97]. For low mass window, $M \lesssim 10^{-12} M_\odot$, the stochastic background of gravity waves emitted during formation of PBHs is expected to be within the sensitivity of the future (2037?) space interferometer LISA, but with the caveat that the astrophysical backgrounds are still highly uncertain (see, e.g., Christensen (2018) in [97]). Similar masses can be tested by detecting the seismic oscillations that a PBH would produce when piercing the body of the Sun or another star (or even the Earth, an event which would, somewhat anticlimactically, only result in a small albeit peculiar earthquake) or neutron stars. Sensing the motion of passing PBHs in the Galaxy via the modification to the timing of the signal of a pulsar could allow to test the $10 - 100 \, M_\odot$ LIGO range. Using pulsar timing arrays (PTA) could probe a much larger mass window: Dror et al. (2019) estimate that future PTA observations with the Square Kilometer Array (SKA) may probe the range $10^{-12} \, M_\odot \lesssim M \lesssim 100 \, M_\odot$; the current NANOGRAV results only constrain $f \lesssim 300$ on the range $10^{-6} \, M_\odot \lesssim M \lesssim 10^4 \, M_\odot$ [98].

## 3.2 Dark Matter as macroscopic objects

DM heavier than the Planck mass $M_{\rm Pl} \approx 2 \, 10^{-5} \, {\rm gram}$ and lighter than the quasi-stable black holes, $M_{\rm Pl}^{5/3}/T_0^{2/3} \sim 10^{15} \, {\rm gram}$, might exist if sub-Planckian particles form composite objects with macroscopic mass [99]. We assume a common mass $M$ and cross section $\sigma$ for scattering on matter, usually rewritten as $\sigma = \pi R^2$. In many models $R$ is the radius of the object, but more generally it can be a parameter of the theory, only alluding to the geometrical meaning. The two parameters $M$ and $R$ then determine the phenomenology. Detecting such candidates is difficult because they are very rare: their flux $\Phi = \rho v / 4\pi M$ is suppressed by the large mass $M$. As a result, their impact rate (in one direction) on a target with area $A$ is

$$\Gamma = \frac{\rho_\odot v A}{4M} \approx \frac{1.6 \, {\rm g}}{{\rm yr}} \frac{A}{M \, {\rm km}^2}, \tag{3.7}$$

where in the numerical estimate we assumed the local DM density $\rho_\odot = 0.4 \, {\rm GeV/cm}^2$ (see eq. (2.11)) and velocity $v = 10^{-3} c$ (see eq. (2.25)). Assuming that in the scattering events the macroscopic object only releases its kinetic energy[11] the bounds, plotted in fig. 3.4, are as follows

- A $\rm m^2$ area inside the Skylab space station has been monitored for a yr, giving the bound $\Gamma \lesssim 1/\,{\rm m}^2 \,{\rm yr}$ that excludes barely sub-Planckian $M$. However, the bound only applies if $\sigma \gtrsim 10^{-18} \, {\rm cm}^2$ (as needed to detectably damage the material) and if $\sigma/M \lesssim 3 \, {\rm cm}^2/{\rm g}$ (as needed to enter the Skylab). The combination of these considerations gives the exclusion region shown in fig. 3.4.

- The lack of tracks in 500 Myr old mica gives the bound $M \gtrsim 55 \, {\rm g}$ if again $\sigma \gtrsim 10^{-18} \, {\rm cm}^2$ (needed to damage mica) and if $\sigma/M < 3 \, 10^{-6} \, {\rm cm}^2/{\rm g}$ (such that DM reached mica underground).

- Similarly, the lack of people killed by impacts with DM gives the bound $M \gtrsim 10 \, {\rm kg}$ if $\sigma \gtrsim 10^{-8} \, {\rm cm}^2$ (needed to deposit as much energy as a bullet) and if $\sigma/M \lesssim 10^{-4} \, {\rm cm}^2/{\rm g}$ (such that energetic DM reaches the ground).

- Cosmology gives the bound $\sigma/M \lesssim 3 \, 10^{-3} \, {\rm cm}^2/{\rm g}$ on dark-matter interactions with baryons, and $\sigma/M \lesssim 5 \, 10^{-7} \, {\rm cm}^2/{\rm g}$ on dark-matter interactions with photons.

- Very heavy DM would produce unseen micro-lensing effects if $M \gtrsim 2 \, 10^{22} \, {\rm g}$ (see section 3.1).

- Macroscopic DM could heat white dwarfs, igniting thermo-nuclear runaways. The existence of long-lived white dwarfs excludes the region in blue in fig. 3.4 [99]. Its complicated shape arises from requiring the DM heating rate to be bigger than the cooling rate, and that DM is able to penetrate the external layers of a white dwarf.

---

[11]More speculative macroscopic objects made of anti-matter would annihilate, releasing more energy.

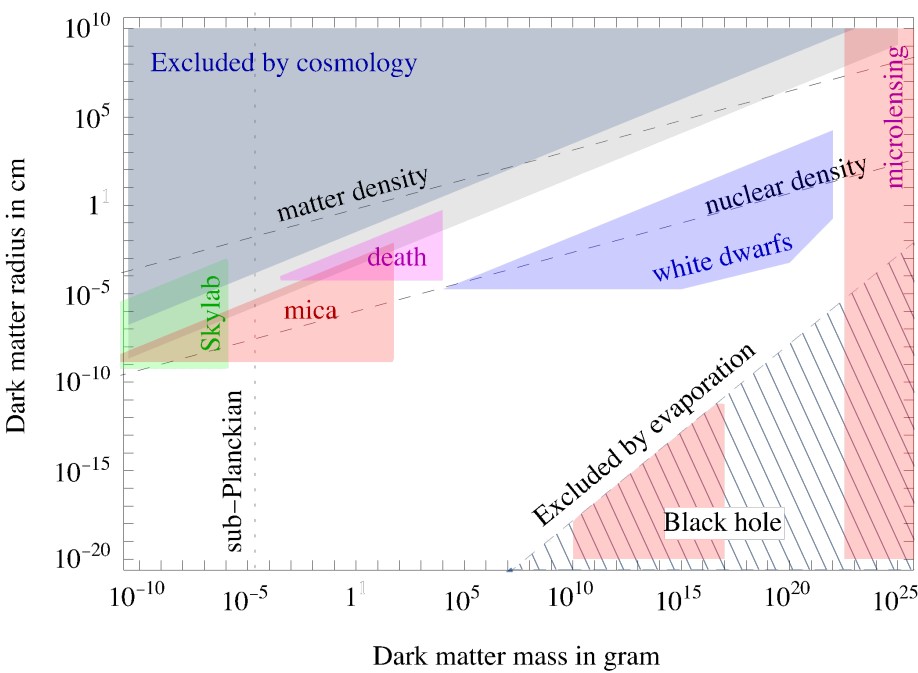

Figure 3.4: *DM is here assumed to be a **macroscopic object** with mass M and radius R, describing a cross section on matter $\sigma \approx \pi R^2$. Shaded regions are bounds from cosmology, Skylab, ancient mica, sudden death by DM, micro-lensing, collapse of white dwarfs, formation of black holes and their evaporation [99]. The dashed lines correspond to object with the density of normal matter or of nuclear matter.*

Possible theories of cosmological formation of DM as macroscopic objects are discussed in section 10.5.

## 3.3   Particle Dark Matter

The most studied possibility is that DM is made of particles. A viable particle DM candidate must have the following properties (rephrasing the properties listed in chapter 1 in particle physics term):

1. DM must be *cold* or at least *not too hot*. Technically, this means that DM needs to be non-relativistic at the time of matter-radiation equality (occurring just before photon decoupling, i.e., at the time of CMB formation, $z \sim 1100$), and henceforth during all the subsequent periods of galaxy formation, as discussed in section 1.3.1. This means that the typical velocity of DM particles, $v$, is much smaller than the speed of light, $v \ll c$, or, equivalently, that their typical momentum, $p$, is much smaller than their mass, $p \ll M$. If DM is made of thermalized particles, this implies that they must be heavier than a few keV, as we discuss in section 3.3.1.

2. The *electric charge* $q$ of a DM particle must be *null or very small*. The experimental bounds on the allowed values reach $q \lesssim 10^{-10}$ for DM with a weak scale mass, $M \lesssim$ TeV. For heavier DM the bounds become gradually weaker, reaching $q \lesssim 0.3$ for Planck mass DM (the bounds are summarized in fig. 3.5). For $q \simeq 10^{-11}$ the freeze-in mechanism (see section 4.2.2) gives the correct relic abundance almost irrespective of DM mass, a possibility that is not experimentally excluded. We expand on some of the details concerning the (milli-)charged DM in section 3.3.2.

| Candidate | Date | Reference(s) | Section |
|---|---|---|---|
| Primordial BH | 1966, 1971 | Zeldovich & Novikov, Hawking [86] | 3.1.1 |
| MACHOs | 1981, 1986 | Petrou; Paczynski [83] | 3.1 |
| Gravitinos | 1981, 1982 | Fayet; Witten; Pagel & Primack [100] | 10.1.2 |
| Axions | 1983 | Preskill, Wise & Wilczek [101] | 10.4 |
| Neutralinos | 1984 | Ellis et al. [102] | 10.1.2 |
| Strangelets | 1984 | Witten; Fahri & Jaffe [103] | 9.1.2 |
| $Q$-balls | 1984 | Witten [104] | 10.5.3 |
| Extra-dimensional DM | 1984 | Kolb & Slansky; Servant & Tait [105] | 10.1.3 |
| WIMPs | 1985 | Steigman & Turner [106] | 9.3.3 |
| Sterile neutrinos | 1993 | Dodelson & Widrow [107] | 9.2.2 |
| Fuzzy DM | 2000 | Hu, Barkana & Gruzinov [108] | 3.4 |
| Sub-GeV DM | 2003 | Boehm, Fayet et al. [109] | 4.1.4 |

Table 3.1: **Some historic DM candidates** *(mostly in the particle DM category) or classes of candidates.*

3. Bounds from direct detection experiments, discussed in section 5, imply that *the cross section for scattering of DM on normal matter is smaller than a typical weak cross section*, $\sigma \ll 1/M_Z^2$, for $M \sim M_Z$. The bound on $\sigma$ becomes much less constraining for $M \ll$ GeV, and becomes less constraining also for $M \gg M_Z$, in which case it scales as $M/M_Z$. Among other things this means that the possible DM coupling to the $Z$ boson must be significantly smaller than a typical weak interaction for DM lighter than $\sim 10^7$ TeV.

4. The *cross section among two DM particles* must be *smaller than a typical QCD cross section*, $\sigma/M \ll (\,\text{GeV}/M)/M_\pi^2$, where $M_\pi = 135$ MeV is the pion mass. This bound is easily satisfied for particles that do not have strong interactions. For larger cross sections DM would not have behaved as a collision-less fluid in cosmology and in astrophysical systems. We discuss the possibility of self-interacting DM in section 3.3.3.

5. The DM particle must be *stable*, or have a lifetime much longer than the age of the Universe ($\approx 13.8$ Gyr $= 4.35 \, 10^{17}$ s) so that the cosmological effects of DM decays are negligible. If DM had decayed too early, it would not have played its important role in shaping the Large Scale Structures of the Universe and/or we would have observed its decay products. Current limits on DM lifetime are of the order of $\tau \gtrsim 10^{28}$ s (see, e.g., fig. 6.15 in section 6.13.2).

In section 9.1 we will see that the above properties are not simultaneously satisfied by any of the Standard Model particles. As a consequence, new unknown and so far undiscovered kinds of fundamental particles are invoked (see table 3.1), as discussed in chapters 9, 10.

Note that the above list does not specify anything about the mass of the DM particle: as long as one does not commit to a specific model or a specific production mechanism, DM particles can be as light as 1 eV or as heavy as $M_{\rm Pl}$ (the edges of the 'particle realm' in eq. 3.1 and in figure 3.2). DM can consist of a very large number of very light particles or, alternatively, of a much smaller number of much heavier particles, or anything in between, where all these possibilities are compatible with all the observations. The allowed range of DM masses does depend, though, on whether the DM particle is a fermion or a boson. Fermionic DM is subject to the Pauli exclusion principle, which leads to a lower bound on DM mass, first derived by **Gunn and Tremaine** [110]. For concreteness, let us consider the Milky Way, where the local DM density is $\rho_\odot \approx 0.4 \, \text{GeV}/\,\text{cm}^3 \approx (0.04 \, \text{eV})^4$ and the galactic escape velocity is $v_{\rm esc} \approx 10^{-3}$. The de Broglie wave-length is $\lambda = 2\pi/Mv$, where $v$ must be smaller than the escape velocity.

Assuming a single DM fermion species, its quantum occupation number is required to be smaller than one. The maximal DM density is therefore $\rho \lesssim M/\lambda^3$, which implies $M \gtrsim 1\,\text{keV}$. Note that throughout this argument we were able to treat DM fermions as free, given the bounds on DM self-interactions (see section 3.3.3). More precisely, a detailed study of dwarf spheroidal galaxies finds [110]

$$M > 0.1\,\text{keV} \qquad \text{(fermionic DM)}. \tag{3.8}$$

This bound is robust and independent from the model of DM formation and DM decoupling. It gets relaxed by a factor $N^{1/4}$, if there are $N$ independent fermion species; bounds on their gravitational production at colliders allow up to $N \lesssim 10^{60}$ [110]. The bound in (3.8) implies, in particular, that the Standard Model neutrinos cannot be the DM, as discussed in more detail in section 9.1.1.

Bosonic DM is not subject to the above bound and can be much lighter. In other words, DM lighter than 0.1 keV must be bosonic. In fact, bosonic DM particles can be so light that this brings us outside the domain of particle DM, as discussed in section 3.4.

## 3.3.1 Warm DM

DM made of particles with negligible velocity is dubbed *cold DM*; *hot DM* refers to quasi-relativistic DM particles, and *warm DM* (WDM) to the intermediate regime [111]. As shown in section 1.3.1, hot DM is excluded by structure formation. SM neutrinos would qualify as hot DM, and thereby cannot be the DM (see section 9.1.1). The intermediate situation is intriguing since the moderate motions of WDM particles smoothen out structures on small cosmological scales (e.g. dwarf galaxies or the cores of galaxies) and could alleviate some of the potential problems with cold DM, see section 8.5. Indeed, this possibility is so interesting that numerical simulations including WDM have been performed by a number of groups [24].

Quantitatively, the observable of interest is the free-streaming length of DM particles at the beginning of structure formation: any structures smaller than this length get washed out. Lyman-$\alpha$ data (i.e. the analysis of absorption inside hydrogen clouds of the light emitted at the Lyman-$\alpha$ frequency from a background source, such as a quasar) allow to probe the distribution of matter on small scales and therefore to test DM free-streaming. Barring possible anomalies, Lyman-$\alpha$ observations agree with CDM and constrain warm DM to be heavier than a few keV: the lighter the DM, the hotter it tends to be. More precisely, this bound is weaker if in the early universe DM was produced via some non-thermal mechanism with energies lower than those of the SM plasma. In principle, the bound depends on the full DM velocity distribution. In practice, however, the bound on the DM mass $M$ can be well approximated using the average momentum-to-temperature ratio $\langle p/T \rangle$, giving [112]

$$M \gtrsim 1.9\,\text{keV} \left\langle \frac{p}{T} \right\rangle_{\text{prod}} \left( \frac{106.75}{g_{\text{SM}}(T_{\text{prod}})} \right)^{1/3}. \tag{3.9}$$

Different groups obtain similar bounds, affected by astrophysical uncertainties. Above, the number of SM degrees of freedom $g_{\text{SM}}$ were normalized to its maximal value, 106.75, which is reached for $T_{\text{prod}} \gg M_Z$. The DM velocity factor $\langle p/T \rangle$ entering (3.9) is determined either by the moment at which DM decouples (if DM was already present in the plasma in thermal equilibrium), or at the moment DM is predominantly produced (out of equilibrium). We mark this moment by the temperature of the SM bath, $T = T_{\text{prod}}$. In the subsequent cosmological evolution the $p/T$ ratio changes only when entropy is injected into the SM plasma, giving at lower temperatures $\langle p/T \rangle = \langle p/T \rangle_{\text{prod}}[g_{\text{SM}}(T)/g_{\text{SM}}(T_{\text{prod}})]^{1/3}$, explaining the form of the last two factors in eq. (3.9).

The DM velocity factor at production equals $\langle p/T \rangle_{\text{prod}} = 3.15$, if DM is a fermionic species that underwent relativistic freeze-out; 3 in the Boltzmann approximation; higher if DM is a sterile neutrino produced via oscillations; 2.45 if DM is produced from decays of a much heavier thermalized scalar;

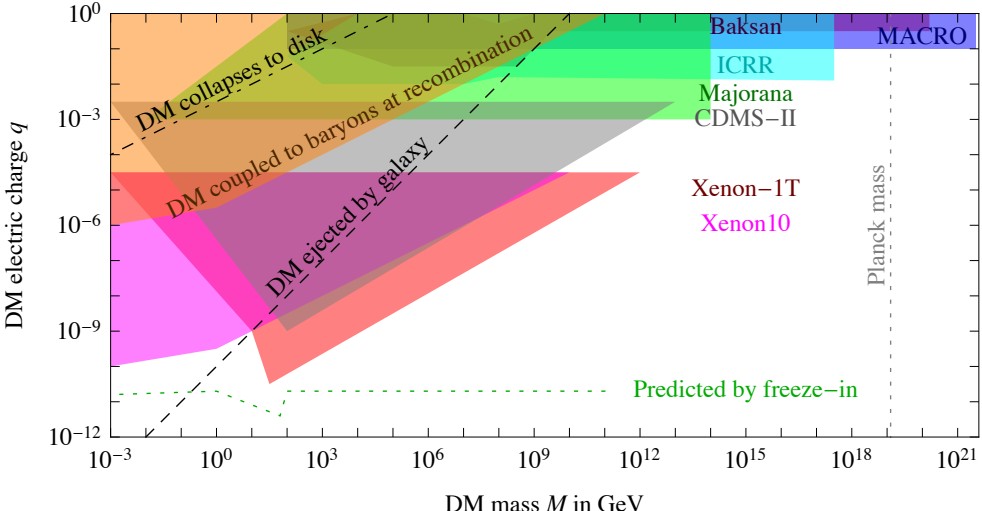

Figure 3.5: *Experimental bounds on a possible **electric charge of DM**. Most of the bounds are reproduced from the references in [113], but some are extended as discussed in the text.*

lower if DM is produced by decays of a slightly heavier particle such that the phase space is partially closed. Bosonic DM can evade the above bounds by developing a classical condensate with negligible $p/T$ (section 3.4).

Lyman-$\alpha$ observations are not the only way to probe the distribution of matter at small scales and therefore test WDM. Another technique involves the observations of strong gravitational lensing events in order to reconstruct the distribution of the intervening matter. One can also consider the number of Milky Way satellites predicted (via numerical or semi-analytical methods) in presence of WDM and compare it with the counting performed by local surveys. The inferred bounds on the DM mass [112] are in the same ballpark as eq. (3.9).

Historically, popular particle physics candidates that behave as warm dark matter include sterile neutrinos (section 9.2.2) and gravitinos (section 10.1.2) [111].

### 3.3.2 (Milli-)charged Dark Matter

A simple possibility for DM to interact with the SM particles is for DM to carry small electromagnetic charge $q|e|$, where $e = -1.6\,10^{-19}$ C is the charge of the electron [114]. This kind of particles are historically refereed to as CHAMPs, for Charged Massive Particles. The charge $e$ is a convenient unit since in plausible contexts, such as the SU(5) gauge unification, uncolored particles must have integer values of $q$ (while for colored particles electric charge is quantized in units of $e/3$).

The possibility $q = 1$ is excluded for all DM particle masses, all the way up to Planck mass. However, $q \ll 1$ may be both phenomenologically viable and theoretically possible. These particles are referred to as *milli-charged DM*, where *milli-* does not necessarily refer to $q \approx 10^{-3}$. Plausible theories can result in much smaller DM charges.[12] A fractional electric charge would make DM stable, preventing its decays

---

[12]DM with a small electric charge can arise if the SM gauge group is extended by a dark U(1) that kinetically mixes with the SM hypercharge $U(1)_Y$, as will be discussed in section 9.4.1. Then $q \sim (\alpha/4\pi)^n$, where $n$ is the number of loops suppressing the kinetic mixing. In models discussed in the literature, up to $n = 4$ loops were found to be needed in models where the abelian factors that mix are part of different non-abelian groups. A higher suppression $q \sim vv'/\Lambda^2 \ll 1$ can arise, if both the hypercharge and the dark U(1) are embedded in non-abelian groups that are broken by vevs $v$ and $v'$, with $\Lambda$ the mass of the heavy states charged under both U(1) factors.

into SM particles.

Fig. 3.5 summarizes experimental bounds on the electromagnetic charge of DM [113]. The scattering of charged DM could lead to signals in various direct detection experiments on the surface of the Earth or deep underground via detectable energy deposits in the form of nuclear recoils, electron recoils, ionization losses or Cherenkov radiation. The scattering cross sections for these processes are proportional to $q^2$. For large enough values of $q$, the scattering cross section becomes so large that non-relativistic charged DM does not reach the underground detectors (resulting in no exclusions from them in fig. 3.5). For lower DM masses a number of these detectors such as CDMS-II, XENON-1T and XENON10 are relevant, and the bounds reach $q \lesssim 10^{-10}$ at around the weak scale, $M \lesssim$ TeV. The bounds on $q$ become weaker with increasing $M$. For ultra-heavy charged DM the best bounds come from reinterpretations of bounds on magnetic monopoles [113]. MACRO was sensitive to slowly-moving ionizing particles with $q \gtrsim 0.1$, and set the bound $\Phi \lesssim 1.4 \ 10^{-16}/\mathrm{cm}^2\,\mathrm{sec\,sr}$ on the number flux of such charged DM particles. Since $\Phi = \rho_\odot v/4\pi M$ this excludes charged DM with $q \gtrsim 0.1$ up to $M \lesssim 5.1 \ 10^{21}$ GeV, above the Planck mass.[13] Another experiment relevant for ultra-heavy charged DM, BAKSAN, was sensitive to $q \gtrsim 0.3$, and set the bound $\Phi \lesssim 2 \ 10^{-15}/\mathrm{cm}^2\,\mathrm{sec\,sr}$, excluding up to $M \lesssim 4\,10^{20}$ GeV [113]. Furthermore, experiments performed at the Institute for Cosmic Ray Research (ICRR) in Tokyo set the bound $\Phi \lesssim 2 \ 10^{-12}/\mathrm{cm}^2\,\mathrm{sec\,sr}$ (thereby excluding charged DM up to $M \lesssim 3 \ 10^{17}$ GeV) but were sensitive down to $q \gtrsim 0.01$ [113]. In deriving the constraints we assumed the usual galactic DM density $\rho_\odot \sim 0.4\,\mathrm{GeV}/\mathrm{cm}^3$ and velocity $v \sim 10^{-3}\,c$, see sections 2.2 and 2.3.

For $M/q \gtrsim rB/v \sim 10^{10}$ GeV (dashed line in fig. 3.5) DM particles are neither appreciably accelerated nor expelled from the Milky Way by supernova (SN) shocks. In this estimate, $v \sim 10^{-3}$ is the virial DM velocity, $r \sim$ pc the rough size of the SN shock and $B \sim 10^{-9}$ T the typical magnetic field inside the shock. For $M/q \lesssim 10^{10}$ GeV the interactions with interstellar medium and SN shocks need to be taken into account since they change the velocity and density profiles of charged DM. Charged DM with $M/q^2 \lesssim 10^5$ GeV (dot-dashed line in fig. 3.5) would virialize with baryons and form a dark disk, contradicting observations (see section 2.4.3).

Weaker bounds come from cosmological observations (and such bounds can be avoided, assuming that only a sub-percent fraction of DM is charged): for $M/q^2 \gtrsim M_{\mathrm{Pl}}\sqrt{T_{\mathrm{rec}}/m_p} \approx 10^{11}$ GeV the Coulomb scatterings of charged DM on baryonic matter at recombination, $T_{\mathrm{rec}} \sim$ eV, would have prevented DM clustering (see e.g. Dolgov et al. (2013) in [113]) such that the CMB acoustic peaks would have changed. Finally, DM with an integer charge would bind with the SM particles and form unusual elements. These are strongly constrained, implying $M \gtrsim 10^7$ GeV [113].[14]

An interesting observation is that for $q \simeq 10^{-11}$ the freeze-in mechanism (section 4.2.2) leads to the observed DM relic abundance [116]. That is, assuming that in the early universe initially only the SM sector was thermalized, the pair production of DM pairs through $s$-channel photon exchanges would lead to the freeze-in abundance

$$\frac{\rho_{\mathrm{DM}}}{s} \simeq 10^{-2}\,\frac{4\pi\alpha^2 q^2}{\sqrt{g_s(M)}}M_{\mathrm{Pl}}, \tag{3.10}$$

where $s$ is the entropy density, and $g_s(M)$ is the effective number of degrees of freedom at $T = M$. The value of $q$ that leads to the correct relic abundance is shown in fig. 3.5 as the green dotted curve. The required value of $q$ is almost independent of $M$ up to $M \lesssim T_{\mathrm{RH}}$, where $T_{\mathrm{RH}}$ is the reheating temperature (see section 4.2.2). For $M$ near the weak scale this scenario could be probed with the future DM direct

---

[13]We thank D. Dunsky and K. Harigaya for the recasting of MACRO results. Note also that Meissner & Nicolai in a series of papers suggest that these constraints may, however, not be robust, in which case an exotic gravitino with Planck mass and charge 2/3 would remain a viable DM candidate [115].

[14] In the standard cosmological evolution most of DM does not form neutral electromagnetic bound states (thereby escaping bounds on charged DM), unless $q/M$ is comparable to its value for hydrogen, a possibility that is in conflict with collider bounds.

detection experiments.

In the simple scenario in which DM with charge $q$ is the only kinematically accessible addition to the SM, the region above the green dotted line in fig. 3.5 is excluded since it predicts too much DM. However, this region is not excluded in non-minimal setups. For instance, the correct relic abundance can be obtained also in the regions above the green line, if DM can efficiently annihilate into other states, e.g., to dark photons. In this case the DM abundance is not set by eq. (3.10), but rather by the annihilation cross section governing the freeze-out process, see section 4.1. The predictions for the relic abundance then become dependent on the details of the model — the couplings of DM to these other states, and which annihilation channels are kinematically allowed. Furthermore, in the non-minimal models of DM with charge $q$ the bounds shown in fig. 3.5 can change and new allowed regions in parameter space open up [117].

### 3.3.3 Self-Interacting Dark Matter

Particle DM might be *Self-Interacting* (SIDM) [118], namely DM particles might undergo elastic scatterings among themselves with a small but nonzero scattering cross section, $\sigma$. The scattering rate

$$\frac{\rho \sigma v_{\rm rel}}{M} \approx \frac{1}{\rm Gyr} \frac{\rho}{0.1 M_\odot/\rm pc^3} \frac{\sigma/M}{\rm cm^2/g} \frac{v_{\rm rel}}{50\,\rm km/\,sec}, \tag{3.11}$$

would be sizeable in dense environments.[15] DM particles in the centers of halos would gain energy by collisions with high-velocity DM particles falling into the gravitational well, and therefore the core of the halo would heat up and become less dense. This effect was in fact even the original motivation for the construction of these types of models, since the formation of cored halos would have lead to an improved agreement between predictions and observations across a wide range of galactic masses, resolving the tension with simulations (see section 8.5.1; there are extensive simulations with SIDM supporting this main observation [24]).[16] The required scattering cross section for the formation of a cored profile to occur is [118]

$$\frac{\sigma}{M} \sim (0.1-1) \frac{\rm cm^2}{\rm g}, \tag{3.12}$$

where $1\,\rm cm^2/\,g = 4578/\,GeV^3$ is a cross-section–to–mass ratio that in the SM is typical for nuclear interactions among hadronic states. Such a relatively large DM-DM scattering cross section can arise in several different ways. For instance, if DM interactions are mediated by a scalar of mass $m$, which couples to DM with coupling strength $g$, then $\sigma = g^4 M^2/4\pi m^4$, and thus for judicial choices of $m$ and $g$ a large enough cross section can be obtained.

Bounds of several different types constrain these scenarios [119]. The observations of colliding galaxy clusters (mainly of the Bullet Cluster, see section 1.2.1) imply $\sigma/M \lesssim 1\,\rm cm^2/g$, because hard collisions would scatter DM particles away [13, 16] and soft frequent collisions, such as those mediated by a light particle, would lead to a drag force that was not observed.

The observations using gravitational lensing indicate that the inner regions of dark matter halos in massive clusters of galaxies and in dwarf galaxies are elliptical in shape. This implies a tighter bound $\sigma/M \lesssim 0.02\,\rm cm^2/g$, because dark matter self-interactions would make halos spherical once most particles have interacted and the momenta have been randomized.

The fact that DM particles in halos of galaxies such as ours do not 'evaporate away' implies an upper bound $\sigma/M \lesssim 0.3\,\rm cm^2/g$: the interactions between DM particles in the galactic halo and those from the halo of the galaxy cluster hosting the Milky Way (these DM particles are on average hotter) would heat

---

[15]Note that $0.1\,M_\odot/\,pc^3$ corresponds to about 10 times the local density (see eq. (2.11)), and is representative of the density in the inner few kpc for a Milky Way like galaxy.

[16](Partially) self-interacting DM has also been used to produce a dark disk, see section 2.4.3.

the MW halo and cause its evaporation, if the scattering cross section is too large. Similar bounds follow from the fact that the halos of dwarfs are not being 'evaporated away' by the MW halo.

Light mediators are constrained by energy losses in supernovæ and by Big Bang Nucleosynthesis. DM models with cross sections that are suppressed at small velocity (e.g. $p$-wave dominated) or models with inelastic DM can, however, alleviate these tensions (see, e.g., Blennow et al. (2017) in [118]).

A related possibility is that DM might be '**radioactive**' [120]:  a bound state that de-excites emitting lighter particles, and thereby also acquiring a recoil energy. The bounds on emissions of $\beta$ or $\gamma$ rays with energies in the MeV-GeV range, however, require lifetimes longer than the age of the Universe.

## 3.4 Dark Matter as waves of light and ultra-light fields

Fermionic DM is subject to the Gunn-Tremaine bound, eq. (3.8), due to the Fermi exclusion principle, while bosonic DM can be lighter and packed tightly together. If we define, as customary, the occupation number $N$ as the ratio between the density $\rho$ of DM particles and the density in a volume associated with their de Broglie wave-length $\lambda = 2\pi/Mv$, for a particle of mass $M$ and velocity $v$, we have

$$N = \frac{\rho}{M/\lambda^3} \approx \left(\frac{30\,\text{eV}}{M}\right)^4 \frac{\rho_\odot}{0.4\,\text{GeV}/\text{cm}^3} \left(\frac{10^{-3}}{v}\right)^3 . \qquad (3.13)$$

Light bosonic DM will thus have $N \gg 1$ and behave effectively as a scalar field.

Provided that the cosmological history features a mechanism that leads to a population of *non-relativistic* bosons with occupation numbers $N$ larger than 1 (e.g., the mechanism known as 'initial misalignment', see below and in section 4.3.4), DM could thus be a light or ultra-light field composed of DM particles with any of the allowed light DM masses, from $M \lesssim$ eV down to the minimum allowed value $M \sim 10^{-21}$ eV (see eq. (3.1)).[17] Being non-relativistic, these very light bosons dilute like matter during cosmological expansion: although the word 'matter' for such a system might appear bizarre, it has the property needed to be an acceptable Dark Matter candidate.

From a theoretical point of view, a scalar field $\varphi$ can be light or ultra-light because it is the pseudo-Goldstone boson of an approximate U(1) global symmetry spontaneously broken at a scale $f$.[18] Periodicity demands that its potential has a trigonometric form $V = \sum_{n=0}^\infty V_n \cos n\varphi/f$ where $V_n$ are constants and the sum runs over different sources of small explicit symmetry breaking. For simplicity we can take

$$V(\varphi) = M^2 f^2 \left(1 - \cos\frac{\varphi}{f}\right) \simeq M^2 \frac{\varphi^2}{2} + \lambda_4 \frac{\varphi^4}{4!} + \cdots, \qquad \lambda_4 = -\frac{M^2}{f^2}, \qquad (3.14)$$

where $M$ is the DM mass. We can often neglect the attractive self-interaction, parametrised by the $\lambda_4$ term. If the spontaneous symmetry breaking occurred at a temperature $T \gtrsim f$, the scalar would typically have an initial value $\varphi_* \sim f$. In general, an initial field value away from the minimum gives a plausible 'misalignment' mechanism for generating the DM cosmological density, to be discussed in section 4.3.4. A pseudo-Goldstone $\varphi$ has interactions of the form $J^\mu(\partial_\mu\varphi)/f$ where $J_\mu$ is the Noether current of a

---

[17]This broad range is allowed on the basis of general principles. In practice, several bounds apply, see figures 6.17 and 10.1.

[18]The spontaneous breaking of an approximate non-abelian global symmetry $\mathcal{G}$ to a subgroup $\mathcal{H}$ results in general in multiple scalars, that parameterise the $\mathcal{G}/\mathcal{H}$ coset. The remnant of the spontaneously broken symmetry is that the Lagrangian for Goldstone bosons has a shift symmetry, i.e., it is symmetric under the change $\varphi \to \varphi + \delta\varphi$. Thus Goldstone bosons only have derivative interactions, up to explicit breaking of the initial symmetry, such as the Goldstone boson mass terms. An exponentially small breaking $M \ll f$ could be generated by instantons, even of gravitational origin, in which case $M \sim M_{\text{Pl}} \, e^{-\mathcal{O}(M_{\text{Pl}}/f)}$ (see, e.g., Alonso & Urbano (2017) [108]).

broken U(1).[19] Interactions with the SM particles could allow to detect such light DM candidates. The phenomenologically most promising is the interaction with photons, and is present in many motivated examples of light and ultra-light DM candidates.

In the next subsections we go over the most popular classes of light and ultra-light DM candidates, which share the broad features discussed above.

## 3.4.1   Light (WISP) Dark Matter

In the upper part of the ultralight/light DM mass range, i.e., for $M \lesssim 1\,\mathrm{eV}$, the light bosons making up DM are known as the **Weakly Interacting Slim Particles (WISPs)**. Several DM candidates fall into this class, including *axions*, *axion-like particles (ALPs)* and *dark* (or *hidden*) *photons*. Historically, axions and axion-like particles have been searched for around $M \sim 10^{-6}$ eV, although higher and lower masses are possible. More recent experiments are broadening the reach of the searches, trying in particular to improve the sensitivity to lighter masses.

The ALPs generalize a notion of axions. The axion mass, $m_a$, and the axion couplings to the SM particles $x = \gamma, e, \ldots$, denoted collectively as $g_{ax}$, are both proportional to the inverse power of the symmetry breaking scale $f_a$, i.e., one has $m_a, g_{ax} \propto 1/f_a$. For axions the $m_a$ and $g_{ax}$ are thus related up to model dependent dimensionless factors (see, e.g., eq.s (10.25), (10.36) and (10.44) below). For ALPs this link is relaxed, and the mass is essentially a free parameter. The phenomenology of the axion models is therefore confined to a band in the $(m_a, g_{ax})$ plane (see, e.g., fig. 10.1), while ALPs can populate the entire plane in this parameter space.

Axions and axion-like particles will be discussed extensively in section 10.4.

## 3.4.2   Ultra-light (fuzzy) Dark Matter

DM with a mass around the lower allowable limit, $M \sim 10^{-21}$ eV, is known as the **fuzzy DM** [108].[20] In the rest of this section we first discuss the field equations for light/ultra-light DM in general, and then focus on the cosmology of fuzzy DM.

Neglecting the cosmological background, the free classical scalar field $\varphi(\boldsymbol{x}, t)$ satisfies the wave equation $\Box^2 \varphi = (\ddot{\varphi} - \nabla^2 \varphi) = -M^2\varphi - \lambda_4 \varphi^3/3!$. In the non-relativistic regime it can be written in terms of the slowly-varying field $\varphi(\boldsymbol{x}, t) = [\psi(\boldsymbol{x}, t)e^{-imt} + \psi^*(\boldsymbol{x}, t)e^{imt}]/\sqrt{2M}$, such that approximating $|\ddot{\psi}| \ll M|\dot{\psi}|$ one gets the Gross-Pitaevskii-Poisson equation

$$i\frac{\partial \psi}{\partial t} = -\frac{\nabla^2 \psi}{2M} + M\phi\psi + \frac{\lambda_4}{8}\frac{f^2}{M}\psi|\psi|^2, \tag{3.15}$$

where we have added on the right side the non-relativistic gravity in the form of a Newton potential $\phi$. In the limit of negligible self-interactions this is a Schroedinger-like equation. The formal analogy with the Schroedinger equation allows one to show that the system behaves similarly to DM. That is, the system can be reformulated as a classical fluid with number density $n = |\psi|^2$ and mass density $\rho = Mn$ up to small gradient terms. Writing $\psi = \sqrt{n}e^{i\theta}$ the velocity field is $\boldsymbol{v} = \boldsymbol{\nabla}\theta/M$. The density and velocity obey

---

[19]For example, the lepton numer $U(1)_L$ could be spontaneously broken by a complex scalar $S = f + (s + i\varphi)/\sqrt{2}$ that couples to right-handed neutrinos through $\mathscr{L}_{\mathrm{int}} \supset S\nu_R^2$. The pseudo-Goldstone boson $\varphi$ is the so called '**majoron**', and couples to a current containing neutrinos, $\bar{\nu}\gamma_\mu\nu$. The majoron, if it acquires a nonzero mass, can be a viable DM candidate [121].

[20]An even lighter scalar, lighter than the present Hubble constant $M \lesssim H_0 \approx 1.6\ 10^{-33}\,\mathrm{eV}$, behaves as the vacuum energy.

the mass conservation equation $\dot{\rho} + \boldsymbol{\nabla} \cdot (\rho \boldsymbol{v}) = 0$ and evolve according to the Euler equation

$$\dot{\boldsymbol{v}} + (\boldsymbol{v} \cdot \boldsymbol{\nabla})\boldsymbol{v} = \boldsymbol{\nabla}\left(-\phi + \frac{1}{2M^2}\frac{\nabla^2\sqrt{\rho}}{\sqrt{\rho}}\right), \tag{3.16}$$

where the latter term, dubbed 'quantum pressure' in view of the mathematical analogy with the quantum Schroedinger equation, accounts for the wave dynamics behind the fluid description. For large $M$ this effect becomes negligible, and the system reduces to DM. More precisely, in cosmology one can expand around small inhomogeneities $\rho = \rho_0(t) + \rho_1(\boldsymbol{x}, t)$ as in eq. (1.7). The divergence of the right-handed side of the Euler equation is $-4\pi G\rho_1 + \nabla^4\rho_1/4M^2\rho_0$. Moving to Fourier space $\nabla^2 \to -k^2$, this allows to compare gravity to 'quantum pressure': the latter dominates at small scales $k > k_J \equiv (M/M_{\rm Pl})^{1/2}(16\pi\rho_0)^{1/4}$. In the present universe $k_J \sim H_0(M/H_0)^{1/2} \simeq 190\,(M/10^{-21}\,{\rm eV})\,{\rm Mpc}^{-1}$.

Fuzzy DM displays an interesting phenomenology, both at cosmological scales and at galactic, stellar and even laboratory scales. The galactic and stellar probes of ultra-light DM are covered in section 6.14, while the laboratory searches are discussed in section 5.8. At the cosmological scales, the most important feature is that the quantum pressure implies that the fuzzy DM leads to a reduced power spectrum of linear inhomogeneities at large $k$, i.e., at small scales. A detailed computation shows that the wavenumber at which the suppression starts is slightly smaller than the $k_J$ introduced above: one obtains $k \sim 12.5\,(M/10^{-21}\,{\rm eV})^{4/9}\,{\rm Mpc}^{-1}$, see Hu et al. (2000) in [108]. Such a suppression of power on small scales is constrained by Lyman-$\alpha$ data: according to various groups, these imply $M \gtrsim 3\,10^{-21}\,{\rm eV}$ [122].

Numerical $N$-body simulations involving fuzzy DM have been successfully performed, despite the difficulty of dealing with the very small resolution of the order of the de Broglie wavelength. They provide hints for the phenomenology discussed above [24].

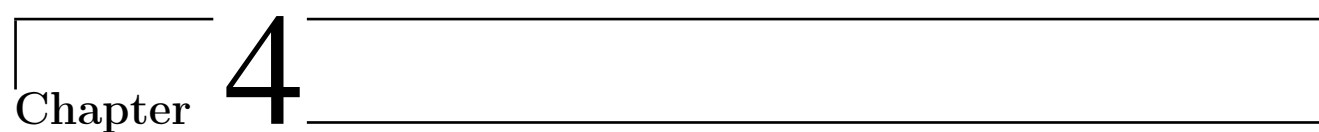

# Chapter 4

# When? Dark Matter production mechanisms

In this section we summarize the main cosmological mechanisms that can produce the observed DM abundance, eq. (1.1).

Section 4.1 discusses DM as a thermal relic (freeze-out). This is often considered to be the most plausible mechanism for DM production, and is certainly the one that has been the most popular in the community for the past few decades. As we will elaborate below, its appeal is twofold: i) the only assumption needed is a 'feebly' interacting stable particle in the primordial thermal bath; ii) this assumption fits well with a number of particle physics theories beyond the Standard Model (BSM), such as SuperSymmetry. The second point is arguably losing some of its attractiveness, because the other, more strongly interacting BSM particles, which are also predicted in such theories, have not yet showed up in any of the searches. The first point, however, still stands strong. Historically, the freeze-out mechanism has been considered predominantly for particles interacting with the weak force of the SM. More recently, however, it has been extended to particles interacting with other (feeble) forces.

Section 4.2 discusses the freeze-in mechanism and some of its variants. As we will see below, this mechanism is quite opposite in spirit to the freeze-out, since it assumes that DM particles are completely absent from the primordial thermal bath. It does, however, also provides a natural paradigm to explain the abundance of DM in terms of a few measurable particle physics properties.

Section 4.3 deals with mechanisms that may play a role in the inflationary phase of the Universe's evolution. In particular, section 4.3.1 discusses DM production directly from the decay of the inflaton field. Section 4.3.4 shows how ultra-light DM can be produced from an initial scalar condensate.

Section 4.4 discusses Asymmetric DM. The attractiveness of this mechanism is that it parallels the presumed production mechanism of ordinary matter, i.e., baryogenesis, thereby potentially providing a natural reason for why the dark and ordinary matter abundances are comparable in size. On the other hand, while many ideas for baryogenesis have been put forward, experimentally, the full set of ingredients for a successful baryogenesis theory has not been yet discovered.

Finally, section 4.5 reviews briefly the DM production mechanisms that feature phase transitions, while section 4.6 describes the production of primordial black hole DM.

Notice that most if not all of the production mechanisms require DM to have some interactions with the ordinary matter and/or with itself, beyond just gravity. These interactions need to be weak enough to satisfy the 'non-interacting' bounds discussed in section 3.3. Still, as we already mentioned above, the existence of such interactions is not required observatonally in any way by the cosmological and astrophysical evidences discussed in chapter 1.

Before presenting the various DM production mechanisms, it is useful to introduce a quantity, which expresses DM density in a form most useful for cosmological computations. The use of number density $n_{\rm DM} = \rho_{\rm DM}/M$ has two 'drawbacks': it depends on the unknown DM mass $M$, and evolves significantly during the cosmological history, i.e., it is a function of time $t$ (or, equivalently, a function of temperature $T$). Ratios such as $n_{\rm DM}/n_\gamma$ evolve less. However, photons do have their own cosmological history, getting periodically heated by the annihilations of charged particles when these become non-relativistic. A more useful quantity to measure the cosmological abundance of DM is instead

$$Y_{\rm DM}(T) \equiv \frac{n_{\rm DM}(T)}{s(T)}, \tag{4.1}$$

where $s = 2\pi^2 g_s(T) T^3/45$ is the total entropy density. Its present day value $s_0$ follows from $T_0 = 2.725\,{\rm K}$ and $g_{s0} = 43/11$.[1] The total entropy in a comoving volume $V$, $S = sV$, is conserved during the cosmological evolution as long as thermal equilibrium holds. The quantity $Y_{\rm DM}(T)$ is therefore very convenient, since it stays constant during most of the cosmological history, whenever the number of DM particles per comoving volume is conserved. This is in particular the case at late times when DM is decoupled, and at early times when DM was possibly in a thermal equilibrium with the SM thermal bath.

Its present day value $Y_{\rm DM0} = Y_{\rm DM}(T_0)$ is linked to the DM mass density $\Omega_{\rm DM}$ as

$$\Omega_{\rm DM} = \frac{\rho_{\rm DM}}{\rho_{\rm cr}} = \frac{s_0 Y_{\rm DM0} M}{3H_0^2/8\pi G} = \frac{688\pi^3 T_0^3 Y_{\rm DM0}\, M}{1485\, M_{\rm Pl}^2\, H_0^2}. \tag{4.2}$$

Inserting the present Hubble constant $H_0 = h \times 100\,{\rm km/sec} \cdot {\rm Mpc}$, we obtain

$$\boxed{Y_{\rm DM0} = \frac{0.44\,{\rm eV}}{M} \frac{\Omega_{\rm DM} h^2}{0.120}}. \tag{4.3}$$

Let us first assume that DM decoupled from the thermal bath while it was still relativistic,[2] which then gives $Y_{\rm DM0} = n_{\rm DM}/s = 45\,\zeta(3)/2\pi^4 \times g_{\rm DM}/g_s(T_{\rm dec})$, with $g_s(T_{\rm dec}) \lesssim 100$ the number of SM degrees of freedom at the time of DM decoupling. Here we are using eq. (C.19) in appendix C, and assuming for definiteness the Fermi-Dirac distributed DM (the bosonic case gives a result that differs just by an $\mathcal{O}(1)$ factor). In this case DM abundance is reproduced for $M \approx 1.6\,{\rm eV}\, g_s(T_{\rm dec})/g_{\rm DM}$, i.e., a thermal freeze-out of relativistic DM results in a hot or warm DM, which is highly disfavored by observations, as discussed in section 3.3.1.[3] In order to obtain cold DM, one needs $M \gtrsim {\rm keV}$, and as a consequence a DM number density that is small or very small: $Y_{\rm DM0} \ll 1$. As discussed in the next section, the non-relativistic decoupling satisfies this requirement.

## 4.1   DM as a thermal relic

The freeze-out mechanism [21,124] assumes that DM is a stable particle with mass $M$, which in the early Universe had interactions with the SM particles that were faster than the Hubble rate. DM was thus a component in the thermal bath, with the SM and DM sectors thermalised to a common temperature $T$,

---

[1]Entropy is discussed in appendix C; the effective number of total degrees of freedom $g_s(T)$ is plotted for the SM in fig. C.2 and equals 106.75 at temperatures much above the weak scale.

[2]The analogous neutrino decoupling is discussed in section 9.1.1.

[3]Relativistic freeze-out of DM can give heavy enough DM, if it happens in a hypothetical 'dark sector' that is decoupled from the SM and also has a lower temperature [123]. This could for instance happen if the inflaton dominantly decays into the SM sector. In such a case, DM could even just be a free particle that does not undergo any freeze-out.

maximising the entropy. In the absence of a conserved DM number, thermalization erases any possible primordial inflationary differences in the two components, so that the same SM/DM fluid filled all the Universe, with common adiabatic inhomogeneities.

DM eventually decoupled: the freeze-out mechanism assumes that the current DM abundance is a left-over (a 'relic') of an incomplete ('frozen') process. Since the iso-curvature inhomogeneities are not re-generated when thermal equilibrium between the SM and DM is lost [125], freeze-out satisfies the observational bound on iso-curvature perturbations in eq. (1.37), which demands the same 'adiabatic' pattern of SM and DM inhomogeneities.

In the next sections we consider freeze-out for various particle-physics processes that might have happened.

## 4.1.1 $2 \leftrightarrow 0$ annihilations: estimate of the relic abundance

Perhaps the most plausible process is annihilations of a pair of DM particles (or annihilations of DM and anti-DM) into SM particles. Introducing a soccer notation we denote this as $2 \leftrightarrow 0$, enumerating explicitly the number of dark-sector particles in the initial and final states, summing over both particles and anti-particles. Any number of SM particles can be present in the initial or final state. Often the dominant process involves two SM particles in the final state:

$$\text{DM DM} \leftrightarrow \text{SM SM}, \qquad \text{where SM denotes any Standard Model particle.} \qquad (4.4)$$

In this section we start sketching the physics involved in this process, providing very rough analytical approximations that will then be refined in sections 4.1.2 and 4.1.3.

The typical particle energy at temperature $T$ is $E \sim T$ and the typical de Broglie wavelength is $1/E$, therefore the number density $n$ and energy density $\rho$ are

$$n \sim T^3, \qquad \rho \sim T^4, \qquad \text{for} \quad T \gg M. \qquad (4.5)$$

When the temperature of the Universe drops below the DM mass, $M$, various processes try to maintain thermal equilibrium. As long as this is maintained, the DM number density follows a Boltzmann-suppressed form

$$n_{\text{DM}} \propto e^{-M/T}. \qquad (4.6)$$

Since DM is stable (or so long-lived that its decays play no role), the dominant process that violates the total DM number (assumed not to be conserved, otherwise see section 4.4) are annihilations involving two DM particles, with total cross section $\sigma$. As the temperature decreases, at some point the DM density becomes so small that the DM DM annihilation rate $\Gamma$ becomes slower than the expansion rate $H$ and the thermal equilibrium can no longer be maintained. At this point DM stops annihilating and its abundance *freezes out*. This happens when

$$\Gamma \sim \langle n_{\text{DM}}\sigma \rangle \lesssim H \sim \frac{T^2}{M_{\text{Pl}}}. \qquad (4.7)$$

On the right hand side, we used $H = \sqrt{8\pi G_N \rho/3}$ with $\rho \sim T^4$, which is valid for radiation domination (more precisely $\rho = \pi^2 g_* T^4/30 \sim T^4$, see Appendix C). For the purposes of these crude estimates, we are neglecting all the numerical factors and keeping only the overall scalings. Using eq. (4.7), the out-of-equilibrium DM relic abundance is conveniently estimated in units of the photon number density $n_\gamma \sim T^3$ as

$$\frac{n_{\text{DM}}}{n_\gamma} \sim \frac{T_{\text{fo}}^2/M_{\text{Pl}}\sigma}{T_{\text{fo}}^3} \sim \frac{1}{M_{\text{Pl}}\sigma M}, \qquad (4.8)$$

where in the last relation we estimated the freeze-out temperature as $T_{\rm fo} \sim M$, determined by the quick drop in scattering rates for $T/M \lesssim 1$ due to the exponential Boltzmann suppression in (4.6). The current DM energy density $\rho_{\rm DM}$ is then

$$\frac{\rho_{\rm DM}}{\rho_\gamma} \sim \frac{M}{T_0}\frac{n_{\rm DM}}{n_\gamma} \sim \frac{1}{M_{\rm Pl}\sigma T_0}. \tag{4.9}$$

Notice the inverse dependence on the annihilation cross section $\sigma$. As a consequence of it, if there are several stable particles, the relic density of the universe is dominated by the one with the smallest annihilation cross section — the weakest particle wins.

As another very crude approximation, let us put $\rho_{\rm DM} \sim \rho_\gamma$ (it should rather be $\rho_{\rm DM} \sim 1000\,\rho_\gamma$ today, see fig. C.1, but again we are only focusing on the overall scalings here), and the typical expression for the annihilation cross section of a particle with mass $M$ mediated by some light particle with dimension-less coupling $g$, $\sigma \sim g^4/4\pi M^2$, we get

$$M/g^2 \sim \sqrt{T_0 M_{\rm Pl}} \sim {\rm TeV}. \tag{4.10}$$

The geometrical average between 'zero' (the present temperature $T_0 \sim 3\,{\rm K}$) and 'infinity' (the Planck mass $M_{\rm Pl}$) turns out to be comparable to the energy reached by the present colliders.

The above rough computation shows that a TeV-scale DM particle interacting with $\mathcal{O}(1)$ couplings results in a DM abundance that is roughly equivalent to the one actually observed. The paradigmatic realization of this kind of DM are WIMPs, Weakly Interacting Massive Particles (see also section 9.3.3): weak-scale DM particles that talk to the SM via $\mathrm{SU}(2)_{\rm L}$ gauge interactions, controlled by the weak couplings $g \sim g_2 \simeq 0.6$, are produced by the thermal freeze-out mechanism in roughly the right amount to explain the observed relic abundance. Historically, this has been dubbed the **'WIMP miracle'**. In passing, we note that, for the case of TeV-scale DM, eq. (4.3) implies $Y_{\rm DM0} \simeq 4\,10^{-12}$ i.e. $n_{\rm DM}/n_\gamma \sim T_0/M \sim 4\,10^{-12}$. This means that DM freeze-out happened at $T_{\rm fo} \sim M/\ln(M/T_0) \sim M/26$, which corrects the above rough approximation.

If DM annihilation is mediated instead by some new boson with mass $M' \gg M$, the interaction becomes an effective dimension-6 operator with coefficient $1/\Lambda^2 \approx g^2/M'^2$, and the cross section is $\sigma \sim M^2/4\pi\Lambda^4$. The correct DM abundance is now reproduced for $M \sim \Lambda^2/{\rm TeV}$ as long as $M/\Lambda$ is still large enough such that the decoupling happens while DM is already non-relativistic, so that $n_{\rm DM} \ll n_\gamma$. As discussed in the next section, the non-relativistic limit allows for a simple precise computation.

### 4.1.2   $2 \leftrightarrow 0$ annihilations: computation of the relic abundance

As it is clear from the discussion in the previous subsection, the hypothesis that DM is a cold thermal relic fixes the DM annihilation cross section. In this subsection we derive its precise value. The standard tool to perform this kind of cosmological computation is the classical Boltzmann equation for the DM number density, $dn_{\mathrm{DM}i}(t,\boldsymbol{x},\boldsymbol{p})/d^3x\,d^3p$ where $i$ is a superindex that denotes DM polarizations, internal degrees of freedom, etc. In the early universe, inhomogeneities are at the $10^{-5}$ level so one can neglect them, i.e., the $\boldsymbol{x}$ dependence in $n$ can be dropped. Furthermore, scatterings that maintain the kinetic equilibrium are fast enough that one can assume a Fermi-Dirac or Bose-Einstein distribution in the DM momentum $\boldsymbol{p}$ (both reduce to the Maxwell distribution in the relevant non-relativistic limit).

Hence, one can write a single equation for the total DM number density $n_{\rm DM}(t)$, summed over all DM polarizations, internal degrees of freedom, etc. This equation has the intuitive form

$$\frac{1}{a^3}\frac{d(n_{\rm DM}\,a^3)}{dt} = \frac{dn_{\rm DM}}{dt} + 3Hn_{\rm DM} = \langle\sigma v_{\rm rel}\rangle(n_{\rm eq}^2 - n_{\rm DM}^2), \tag{4.11}$$

where $a(t)$ is the scale factor of the universe, $H = \dot{a}/a$ is the Hubble rate, $n_{\rm eq}$ is the number density that DM particles would have in thermal equilibrium and $\langle\sigma v_{\rm rel}\rangle$ is the thermal average of the total annihilation cross section times the relative velocity $v_{\rm rel}$ between annihilating DM particles. The term $3Hn_{\rm DM}$ in the

expression after the first equality accounts for the dilution of DM density due to the expansion of the Universe. The two contributions on the right hand side describe respectively the DM depletion due to annihilations (proportional to $n_{\rm DM}^2$) and the DM creation via inverse annihilations (proportional to $n_{\rm eq}^2$). Provided that $\langle\sigma v_{\rm rel}\rangle$ is large enough, the right hand side expression ensures that DM abundance is in a thermal equilibrium, i.e., $n_{\rm DM} = n_{\rm eq}$. As the temperature decreases, DM becomes non-relativistic and its equilibrium density is exponentially suppressed: $n_{\rm eq} = g_{\rm DM}(MT/2\pi)^{3/2}e^{-M/T}$. Here, $g_{\rm DM}$ is the number of degrees of freedom associated with the DM particle ($g_{\rm DM} = 1$ for real scalar DM, $g_{\rm DM} = 2$ for Majorana fermion DM, etc, see fig. 4.1 right). Eventually, the DM density becomes so suppressed that the annihilation term fails to keep up with the expansion term (the DM particles are so rare that they are not able to find each other to annihilate), annihilations stop and thus DM decouples. The observed cosmological abundance of DM is reproduced if thermal equilibrium fails and DM freeze-out occurs at $T < T_{\rm fo} \sim M/25$, such that $n_{\rm DM} \approx n_{\rm eq}(T = T_{\rm fo})$. An accurate approximation to the analytic solution of the Boltzmann equation [124] will be derived in section 4.1.3, with the final result given by

$$\frac{\Omega_{\rm DM}h^2}{0.110} \approx \frac{\sigma v_{\rm cosmo}}{\langle\sigma v_{\rm rel}\rangle_{T\approx M/25}}, \tag{4.12}$$

where

$$\sigma v_{\rm cosmo} \approx 2.2 \times 10^{-26}\,\frac{\rm cm^3}{\rm s} = \frac{1}{(23.0\,{\rm TeV})^2} \approx 0.73\,{\rm pb} = 0.73\,10^{-36}\,{\rm cm}^2. \tag{4.13}$$

The precise numerical result for the special value $\sigma v_{\rm cosmo}$, which reproduces the cosmological DM abundance, is plotted in fig. 4.1 as a function of $M$. Another way of stating the 'WIMP miracle' is that the required $\sigma v_{\rm cosmo}$ turns out to 'miraculously' correspond to a typical weak annihilation cross section, $\sigma \sim g_2^4/4\pi\,{\rm TeV}^2$. In more detail, however, this 'miracle' does need a TeV energy, i.e., an order of magnitude above $M_{W,Z,h} \sim 100\,{\rm GeV}$.

Notice that $\sigma$ is averaged over the various DM components. This leads to factors of 2 that can generate confusion. If DM is a real particle (e.g. a Majorana fermion) $\sigma$ is given by the usual definition of a cross section — averaged over initial polarisations and summed over final states. A factor of 2 on the right hand side of eq. (4.11), present because 2 DM disappear in each annihilation, is canceled by the initial state symmetry factor $1/2$.

If DM is a complex particle (e.g. a Dirac fermion) then $n \equiv n_{\rm DM} + n_{\overline{\rm DM}}$. Here we assume that there is no asymmetry, so that $n_{\rm DM} = n_{\overline{\rm DM}}$. The average over the 4 possible initial states is $\sigma \equiv \frac{1}{4}(2\sigma_{\rm DM\,\overline{\rm DM}} + \sigma_{\rm DM\,DM} + \sigma_{\overline{\rm DM}\,\overline{\rm DM}})$. In many models only DM$\overline{\rm DM}$ annihilations are present, giving $\sigma = \frac{1}{2}\sigma_{\rm DM\,\overline{\rm DM}}$. This discussion can be extended, to deal with more complicated situations where multiple DM states are present, e.g., degenerate weak-multiplets containing both real and complex components. Concrete examples are given in section 9.

### 4.1.3   $2 \leftrightarrow 0$ annihilations: analytic approximation

In this section we address in more detail the description of the freeze-out process that we qualitatively discussed above.[4] For this purpose, it is convenient to a) study the evolution of the total DM abundance

---

[4] In the computations of this section we always assume that, thanks to DM-SM collisions, DM particles remain in kinetic equilibrium with the SM particles down to temperatures that are (a few orders of magnitude) below the thermal decoupling temperature of DM annihilations. This means that we can then neglect the details of the DM energy distribution and concentrate on the integrated quantity, the total DM number density. If, for any reason, kinetic equilibrium does not hold, then the standard computations are affected (see, e.g., Binder et al. (2017) and Binder et al. (2021) [63]).

In the rest of this footnote, let us review the basic concepts of the different equilibriums and decouplings. The term **thermal (or chemical) equilibrium** refers to the equilibrium between particle species. It is maintained

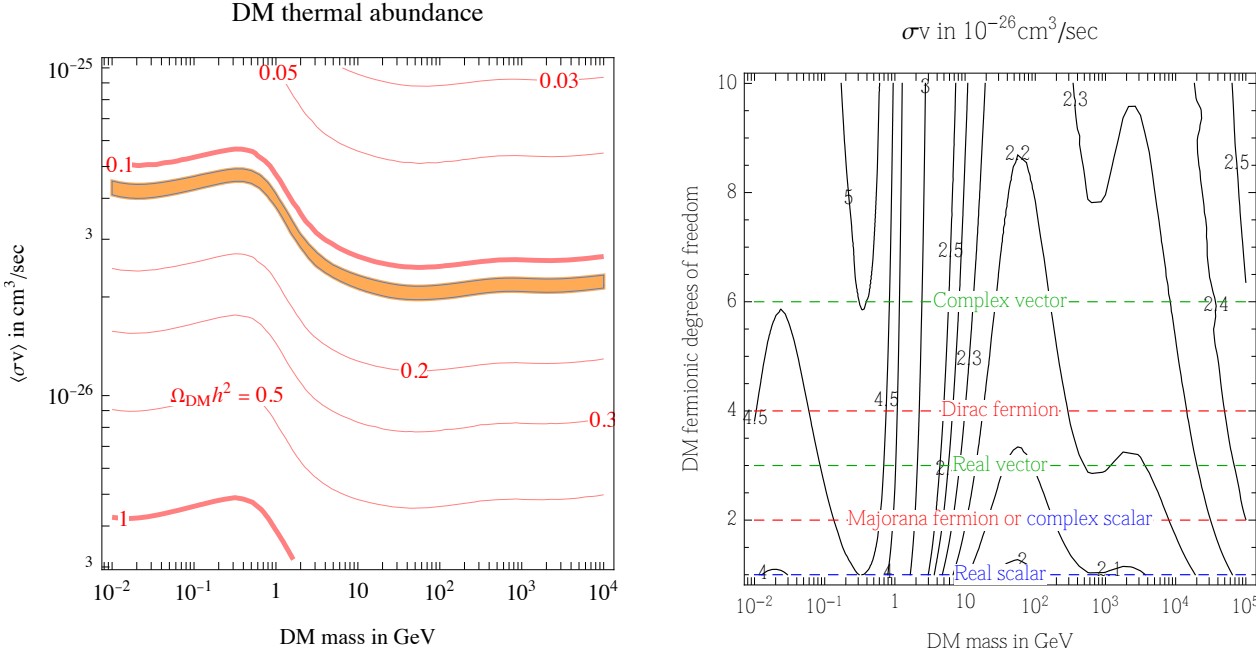

Figure 4.1:   **Left**: *DM freeze-out abundance $\Omega_{\mathrm{DM}}h^2$ as function of the DM mass and of the DM annihilation cross section $\langle \sigma v \rangle$ for Majorana DM. The measured cosmological DM abundance $\Omega_{\mathrm{DM}}h^2 = 0.1200 \pm 0.0012$, eq. (1.1), is reproduced within the orange band at 3 standard deviations. For smaller (larger) annihilation cross sections, i.e. in the lower (upper) part of the plot, the freeze-out happens too early (too late) and therefore the DM relic abundance is too high (too low). **Right**: Iso-contours of the annihilation cross section $\sigma v_{\mathrm{rel}}$ in $10^{-26}$ cm$^3$/sec that reproduces the central value of the DM cosmological abundance as function of the DM mass and of the number of DM degrees of freedom. This cross-section is a target for indirect DM detection searches.*

as a function of a dimensionless parameter $z \equiv M/T$, which is $\mathcal{O}(1)$ during freeze out[5] and which grows with time as $T$ decreases; b) factor out the overall expansion of the universe by writing equations for

by the balance between DM annihilations into SM particles and the inverse process of SM particles creating DM. When thermal equilibrium fails, DM particles *thermally* (or *chemically*) *decouple* from the thermal bath, i.e,. they freeze-out, as extensively discussed above. **Kinetic equilibrium** [63] refers to the equilibrium in the energy distributions. It is maintained by very frequent elastic scatterings between DM and SM particles (in particular, with the dominant SM radiation fluid), and typically holds even at temperatures well below $T_{\mathrm{fo}}$. When the kinetic equilibrium fails, DM particles *kinetically decouple*, therefore their final kinetic distribution in general differs from that of the SM particle species. The temperature of kinetic decoupling, $T_{\mathrm{kd}}$, can be estimated by imposing $\Gamma_{\mathrm{kin}} \sim H$ where $H \sim T^2/M_{\mathrm{Pl}}$, and $\Gamma_{\mathrm{kin}} \sim \sigma n T/M$ is the rate of DM energy exchange due to scatterings with cross section $\sigma$ on SM particles with number density $n$. Assuming $n \sim T^3$ (possibly electrons and neutrinos) and $\sigma \sim g^2 g'^2_{\mathrm{DM}} T^2/M'^4$, where $M'$ is the mass of some mediator that couples with $g$ and $g'_{\mathrm{DM}}$ couplings to matter and DM currents, respectively, one obtains

$$T_{\mathrm{kd}} \sim (M/M_{\mathrm{Pl}})^{1/4} M'/\sqrt{g\, g'_{\mathrm{DM}}}. \qquad (4.14)$$

The temperature of kinetic decoupling is higher (i.e., the decoupling is earlier) the heavier the DM and the mediator are, and the smaller the couplings. For WIMP DM, $M \sim$ TeV, $M'$ is the mass of the $Z, W^\pm$ bosons and $g = g'_{\mathrm{DM}} = g$, so $T_{\mathrm{kd}} \simeq 10$ MeV, a few orders of magnitude below the DM freeze-out temperature $T_{\mathrm{fo}} \sim M/25$.

[5]Or rather $\mathcal{O}(25)$ for weak-scale particles, as seen above.

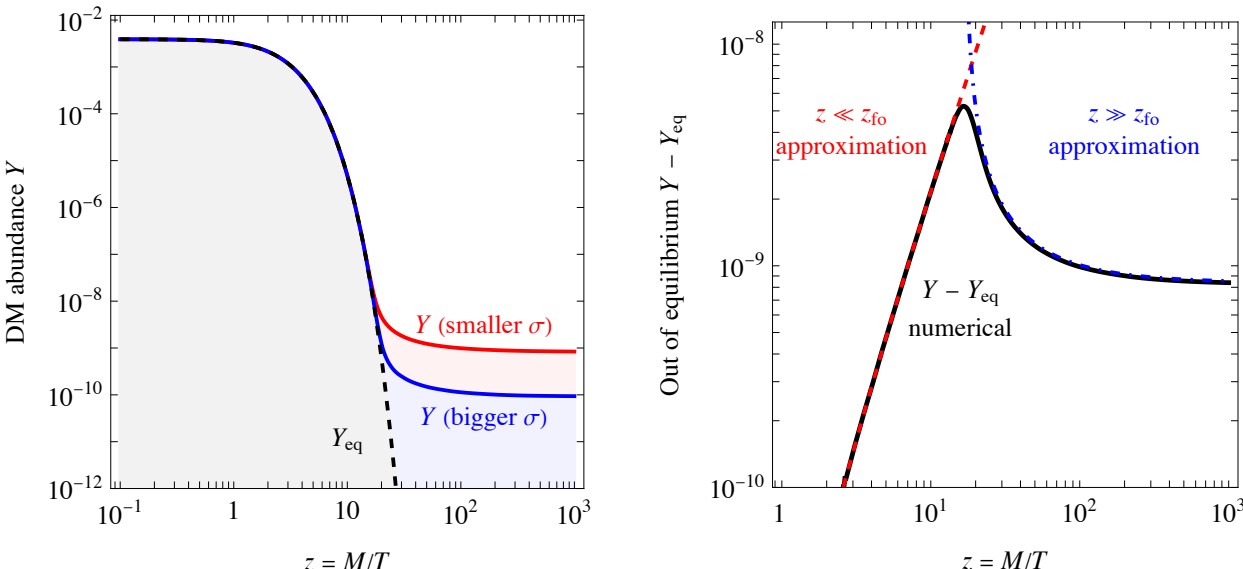

Figure 4.2: *DM freeze-out examples.* **Left:** *sample of the evolution of the DM abundance $Y = n/s$ as function of $z = T/M$ for two different values of the DM annihilation cross section $\sigma$, compared to the equilibrium abundance $Y_{\mathrm{eq}}$ (dashed).* **Right:** *sample of the evolution of the non-equilibrium abundance $Y - Y_{\mathrm{eq}}$, compared to the analytic approximations in eq. (4.20) (dashed, valid for $z \ll z_{\mathrm{fo}}$) and in eq. (4.22) (dot-dashed, valid for $z \gg z_{\mathrm{fo}}$).*

$Y_{\mathrm{DM}} = n_{\mathrm{DM}}/s$, the DM abundance in units of the entropy density introduced in eq. (4.1). The Boltzmann equation in terms of $Y_{\mathrm{DM}}(z)$ is (for details see appendix C.5)

$$sHZz\frac{dY}{dz} = -2\gamma_{\mathrm{ann}}\left(\frac{Y^2}{Y_{\mathrm{eq}}^2} - 1\right), \tag{4.15}$$

where the factor $Z = 1/(1 + \frac{1}{3}\frac{d\ln g_{*s}}{d\ln T})$ can often be approximated as 1, and $\gamma_{\mathrm{ann}}$ is the space-time density of annihilations in thermal equilibrium, summed over initial state and final states and their polarisations. Eq. (C.38) gives $\gamma_{\mathrm{ann}}$ in terms of the annihilation cross section $\sigma(s)$. One can easily perform numerical calculations or run public codes such as `MicrOMEGAs` [126], `DarkSUSY` [127], `MadDM` [128], `SuperIsoRelic` [129] or others.

Below we show instead how one can obtain precise analytic approximations, which are then compared with the numerical solutions in fig. 4.1b. We assume that DM annihilates into lighter particles, so that this process is allowed also at very low temperatures. In the non-relativistic limit, the expression for the annihilation rate simplifies to

$$2\gamma_{\mathrm{ann}} \overset{T \ll M}{\simeq} n_{\mathrm{eq}}^2 \langle \sigma v_{\mathrm{rel}} \rangle, \tag{4.16}$$

where $\langle \sigma v_{\mathrm{rel}} \rangle$ is the thermal average of $\sigma v_{\mathrm{rel}}$. In the non-relativistic limit the annihilation cross section can be expanded in powers of $v_{\mathrm{rel}} \ll 1$ as $\sigma v_{\mathrm{rel}} \simeq \sigma_0 + v_{\mathrm{rel}}^2 \sigma_1 + \cdots$. The leading $\sigma_0$ term is known as **s-wave** and the sub-leading $v_{\mathrm{rel}}^2 \sigma_1$ term as **p-wave**. The thermal average is

$$\langle \sigma v_{\mathrm{rel}} \rangle = \sigma_0 + \frac{6T}{M}\sigma_1 + \cdots. \tag{4.17}$$

The Boltzmann equation for the total DM abundance then simplifies to

$$\frac{dY}{dz} = -f(z)(Y^2 - Y_{\rm eq}^2), \qquad f(z) \equiv \frac{s\langle\sigma v_{\rm rel}\rangle}{HZz} \overset{Z=1}{\approx} \frac{\lambda}{z^2}\left(1 + \frac{6\sigma_1}{z\sigma_0}\right), \tag{4.18}$$

where $Y_{\rm eq} \equiv n_{\rm eq}/s \overset{z\gg1}{\simeq} g_{\rm DM}45z^{3/2}e^{-z}/g_{\rm s}2^{5/2}\pi^{7/2}$, having used $n_{\rm eq}$ from eq. (C.29). The dimension-less constant $\lambda$ is

$$\lambda = \left.\frac{s\langle\sigma v_{\rm rel}\rangle}{H}\right|_{T=M} = M_{\rm Pl}M\langle\sigma v_{\rm rel}\rangle\sqrt{\frac{\pi\,g_{\rm s}^2}{45\,g_\rho}} \gg 1. \tag{4.19}$$

The estimate $Y \sim 1/f|_{T\sim M}$ of eq. (4.8) is recovered by neglecting the inverse-annihilation contribution $Y_{\rm eq}^2$ and writing eq. (4.18) as $d\ln Y/dz \sim -fY$. Furthermore, we can derive the following approximate solutions, plotted in fig. 4.2:

◁ Long before freeze-out, i.e., at early $z \ll z_{\rm fo}$, $Y(z)$ follows $Y_{\rm eq}(z)$ closely, so that one can expand to the first order in their small difference: explicitly, one recasts eq. (4.18) in terms of $Y - Y_{\rm eq}$, neglects the $(Y - Y_{\rm eq})^2$ higher order term, as well as the $(Y - Y_{\rm eq})'$ term (here $'$ denotes the derivative with respect to $z$). One then finds an approximate solution for $Y$:

$$Y(z) \overset{z\ll z_{\rm fo}}{\simeq} Y_{\rm eq} - \frac{Y_{\rm eq}'}{2fY_{\rm eq}} \overset{z\gg1}{\approx} Y_{\rm eq} + \frac{z^2}{2\lambda}, \tag{4.20}$$

where the last relation follows from applying the derivative on $Y_{\rm eq}(z)$ and recalling that we are always in the regime $z \gg 1$. Next, we identify the moment of freeze-out, $z_{\rm fo}$, at which $Y(z)$ ceases to track $Y_{\rm eq}$. We define this to be the moment when the discrepancy $Y - Y_{\rm eq}$, which is just $z^2/2\lambda$ according to eq. (4.20), becomes equal to $Y_{\rm eq}$ itself. This gives the equation

$$z_{\rm fo} = \ln\frac{2e\,\lambda Y_{\rm eq}(1)}{\sqrt{z_{\rm fo}}}, \tag{4.21}$$

which can be iteratively solved starting from $z_{\rm fo} \approx \ln\lambda Y_{\rm eq}(1) \approx 1/25$ (here $Y_{\rm eq}(1)$ is just a convenient short-hand way of writing the numerical coefficients in $Y_{\rm eq}(z)$).

▷ Long after freeze-out, i.e., at late $z \gg z_{\rm fo} \approx 25$, we can neglect the $Y_{\rm eq}^2$ term in eq. (4.18), giving a differential equation, $dY/dz = -fY^2$, which can be easily solved,

$$\frac{1}{Y_\infty} - \frac{1}{Y(z)} \overset{z\gg z_{\rm fo}}{\simeq} \int_\infty^z f(z)\,dz = \frac{\lambda}{z}\left(1 + \frac{3\sigma_1}{z\sigma_0}\right). \tag{4.22}$$

Since $Y(z_{\rm fo}) \gg Y_\infty$ we have the approximate solution

$$Y_\infty = \frac{z_{\rm fo}\sqrt{45g_\rho/\pi g_{\rm s}^2}}{M_{\rm Pl}M(\sigma_0 + 3\sigma_1/z_{\rm fo})}. \tag{4.23}$$

If after freeze-out nothing in the evolution of the Universe changes the entropy in the visible sector, $Y_\infty$ can be identified with the present DM entropy density, $Y_{\rm DM0}$. Converting it into the DM mass density via eq. (4.2) we obtain the precise version of the result announced in eq. (4.12),

$$\frac{\Omega_{\rm DM}h^2}{0.110} = \frac{Y_\infty M}{0.40\,{\rm eV}} = \frac{z_{\rm fo}}{25}\frac{2.18\ 10^{-26}{\rm cm}^3/{\rm s}}{\sigma_0 + 3\sigma_1/z_{\rm fo}}. \tag{4.24}$$

Fig. 4.1b shows the precise value of $\sigma v_{\rm rel}$ that reproduces the central value of DM cosmological abundance as a function of DM mass and of the number of DM degrees of freedom. The somewhat

counterintuitive result is that in these equations the number of degrees of freedom in the DM system only enters with weak logarithmic dependence implicit in $z_{\rm fo}$. Note that $z_{\rm fo} \approx 25$ also means that a thermal relic DM has to be heavier than $M \gtrsim 10\,{\rm MeV}$. Lighter DM would have been in thermal equilibrium with the SM at $T \sim {\rm MeV}$, and this would have increased the Hubble rate enough to conflict with the successful SM big-bang-nucleosynthesis results.

### 4.1.4 Sample cross sections and the unitarity limit on the DM mass

The treatment above is generic and model independent. The final result, eq. (4.24), shows that the annihilation cross section $\sigma v_{\rm rel}$ is the crucial parameter that determines the DM relic abundance. Here we provide sample computations of annihilation cross sections in generic models, in order to make contact with the particle physics model parameters (essentially, couplings and masses).

We consider a DM particle in a representation $R$ of an unbroken generic gauge group $\mathcal{G}$. The non-relativistic $s$-wave tree-level annihilation cross section into massless vectors is given by

$$\sigma_0 = \frac{2C_R(C_R - C_G/4)}{g_{\rm DM}}\frac{\pi\alpha^2}{M^2}, \tag{4.25}$$

where $\alpha = g^2/4\pi$ is the gauge coupling, $C_R$ is the quadratic Casimir defined in terms of generators as $\delta_{ij}C_R = (T^aT^a)_{ij}$; $g_{\rm DM}$ is the number of DM degrees of freedom, equal to $d_R$ for a real scalar, $2d_R$ for a complex scalar or for a Majorana fermion, $4d_R$ for a Dirac fermion. Here $d_R$ is the dimension of the representation. The values of $d_R$ and $C_R$ in the adjoint representation are denoted as $d_G$ and $C_G$. For $\mathcal{G} = {\rm SU}(2)$ one has $d_G = 3$, $C_R = (d_R^2 - 1)/4$ for a generic representation with dimension $d_R$. For $\mathcal{G} = {\rm SU}(3)$ one has $d_G = 8$, $C_3 = 4/3$, $C_8 = 3$, $C_{10} = 6$, $C_{27} = 8$, etc. For $\mathcal{G} = {\rm SU}(N)$ one has $d_G = N^2 - 1$, $C_N = (N^2 - 1)/2N$, $C_{d_G} = N$.

More in general, a typical $s$-wave $2 \to 2$ DM annihilation cross section can be estimated as $\langle \sigma v_{\rm rel}\rangle = \sigma_0 \approx g^4/4\pi M^2$ where $g$ is some unspecified dimension-less DM coupling, such as a gauge or Yukawa coupling. Inserting this estimate in eq. (4.24) implies that the DM mass which reproduces the cosmological DM abundance, is given by $M \approx g^2 \times 6.5\,{\rm TeV}$. The perturbativity upper bound $g^2 \lesssim 4\pi$ then implies an upper bound on the DM mass:

$$\boxed{M \lesssim 100\,{\rm TeV}}. \tag{4.26}$$

This is known as the **unitarity bound** [130].[6] Its rigorous version for the $s$-wave cross section is $\sigma v_{\rm rel} \leq 4\pi/M^2 v$. However, waves with higher $\ell$ give extra contributions, such that in general no rigorous upper bound exists:

$$\sigma v_{\rm rel} \leq \frac{4\pi}{M^2 v_{\rm rel}}\sum_{\ell=0}^{\infty}(2\ell + 1) = \infty. \tag{4.27}$$

Indeed, bound states and more general classical objects with size $R$ have large geometrical cross sections $\sigma \sim \pi R^2$ after summing partial waves up to $\ell \lesssim M v_{\rm rel} R$, which is the classical angular momentum. Model-dependent computations that try to determine which $\ell$ contribute by computing tree-level scatterings do not achieve a higher precision than estimates, given that the tree-level approximation fails precisely when $g^2 \sim 4\pi$. Long-range forces can saturate the unitarity bound, as discussed in section 4.1.5.

At the other end of the spectrum, a lower bound on DM mass can be derived from the requirement of reproducing the cosmological DM abundance via annihilations. Assuming a DM particle that annihilates via the SM weak interactions through an $s$-wave process, the annihilation cross section is estimated as $\sigma_0 \approx G_{\rm F}^2 M^2/2\pi$, where $G_{\rm F} = \sqrt{2}g_2^2/8M_W^2$ is the Fermi constant. Inserting this estimate in eq. (4.24)

---

[6]The often quoted bound $M \lesssim 340\,{\rm TeV}$ corresponds to the looser requirement $\Omega_{\rm DM}h^2 < 1$, as in the original study [130].

implies $M \gtrsim 3\,\mathrm{GeV}$ in order to avoid a too large freeze-out DM relic density (i.e. in order to have $\Omega_{\mathrm{DM}}h^2 \lesssim 0.110$). This is known as the **Lee-Weinberg bound** [131]. Of course, this bound does not apply if DM has a different annihilation cross section, not due to the SM weak interactions.[7] Together with the more general eq. (4.26), it does define, however, the ballpark mass range for WIMP DM.

### 4.1.5   Sommerfeld and bound-state corrections

Non-relativistic cross-sections can receive large non-perturbative corrections, known as the Sommerfeld corrections. These effects tend to grow at low velocities, enhancing DM annihilations, and are thus relevant for cosmology (thermal freeze-out) and indirect detection (section 6), since in both processes DM is non-relativistic. Furthermore, since in galaxies DM has a lower typical velocity than it had during the freeze-out, the annihilation cross section in indirect detection can be above the standard freeze-out value, eq. (4.13).

A classical analogy can help understand the physics: the cross section for falling into the sun (radius $R$, mass $M_\odot$) equals the geometric area $\pi R^2$ only for a body with a velocity much larger than the escape velocity $v_{\mathrm{esc}} = \sqrt{2GM_\odot/R}$. At lower velocities the gravitational attraction increases the cross section to $\sigma = \pi R^2(1 + v_{\mathrm{esc}}^2/v_{\mathrm{rel}}^2)$, where $v_{\mathrm{rel}}$ is the initial velocity of an infalling body (see also below). A repulsive interaction would instead make the cross section smaller.

In the quantum regime, the non-relativistic cross sections for $ee$ scattering and $e\bar{e}$ annihilation get distorted by the long-range electromagnetic interaction when $v_{\mathrm{rel}} \lesssim e^2$, i.e., in the regime where the kinetic energy $\sim m_e v_{\mathrm{rel}}^2$ is smaller than the potential energy $e^2/r$ for separations comparable to the de Broglie wave-length, $r \sim 1/m_e v_{\mathrm{rel}}$.

While DM has no electric charge and gravity is irrelevant at sub-Planckian energies, the Sommerfeld corrections can affect the non-relativistic DM annihilations [132], if DM couples with dimension-less coupling $g$ to a mediator particle with small mass, $M_V \ll M$. Such a light particle mediates what is effectively a long-range force relevant both in cosmology and in indirect detection. Such particle could be an electroweak vector gauge boson, if $M$ is well above the electroweak scale, or a new speculative 'dark' vector coupled to DM, or a scalar with a Yukawa (trilinear) coupling to fermionic (scalar) DM.

The Sommerfeld correction factor $S$, defined through $\sigma = S\sigma_{\mathrm{pert}}$ where $\sigma_{\mathrm{per}}$ is the perturbative (most often tree-level) cross section, can be computed from the distortion of the wave function $\psi(\boldsymbol{r})$ of the two-body DM-DM (or DM-$\overline{\mathrm{DM}}$) initial state relative to the plane wave. To simplify the discussion let us focus on $s$-wave annihilation, and assume, at first, a massless mediator, giving a Coulomb like potential $V = -\alpha/r$ between the two DM particles. Defining $\psi(\boldsymbol{r}) = u(r)/\sqrt{4\pi}r$, the function $u(r)$ satisfies the Schrödinger equation $-u''/M - \alpha u/4\pi r = Eu$, with the outgoing boundary condition $u'(\infty)/u(\infty) \simeq iMv_{\mathrm{rel}}/2$.[8] The Sommerfeld factor is then given by $S = |u(\infty)/u(0)|^2$, and is

$$S = \frac{2\pi\alpha/v_{\mathrm{rel}}}{1 - e^{-2\pi\alpha/v_{\mathrm{rel}}}}. \tag{4.28}$$

---

[7] Historically, the Lee-Weinberg bound presented a conceptual barrier against considering thermally-produced DM lighter than a few GeV. At the time, the dominant paradigm was a DM, typically fermionic, which annihilates via SM weak interactions. This scenario was motivated by supersymmetry and naturalness considerations (see section 10.1.2), and the Lee-Weinberg bound applies to it directly. In the early 2000s, however, it was realized that, if DM particles annihilate via other forces rather than the SM weak interactions, then the bound can be circumvented, allowing for lighter DM to still be thermally produced in the right amount without overclosing the Universe. This conceptual breakthrough initiated the theoretical explorations of **sub-GeV DM** [109]. The models of sub-GeV DM typically feature a dark photon or other light dark force mediators (see section 9.4.1 for further details, as well as section 4.1.9 for production mechanisms specific to sub-GeV DM, sections 5.2, 5.3 and 5.5.4 for direct detection of sub-GeV DM, section 6.13.2 for indirect detection, and section 7.4 for accelerator searches).

[8] The Sommerfeld correction could also be computed using incoming boundary conditions.

In the attractive case (corresponding to $\alpha > 0$) $S$ grows for $v_{\rm rel} \to 0$ roughly as $S \propto v_{\max}/v_{\rm rel}$ for $v_{\rm rel} < v_{\max} = \pi\alpha$. Note that this growth is still consistent with the unitarity bound in eq. (4.27).

If the mediator has mass $M_V$, an analytic solution for $S$ can still be obtained by approximating the Yukawa potential with the Hulthen potential

$$\frac{e^{-M_V r}}{r} \approx \frac{\kappa\, M_V e^{-\kappa M_V r}}{1 - e^{-\kappa M_V r}}, \tag{4.29}$$

where the approximation is most accurate for $\kappa \approx 1.74$. The Sommerfeld factor is then

$$S = \frac{2\pi\alpha \sinh\left(\pi M v_{\rm rel}/\kappa M_V\right)}{v_{\rm rel}\left[\cosh\left(\pi M v_{\rm rel}/\kappa M_V\right) - \cosh\left(\pi M v_{\rm rel}\sqrt{1 - 4\alpha\kappa M_V/M v_{\rm rel}^2}/\kappa M_V\right)\right]}, \tag{4.30}$$

and depends only on $\alpha/v_{\rm rel}$ and $\kappa M_V/M\alpha$. In the case of an attractive potential, $S$ grows roughly as $v_{\max}/v_{\rm rel}$ in the range $M_V/M = v_{\min} < v_{\rm rel} < v_{\max} = \pi\alpha$, and is constant at smaller $v_{\rm rel}$. The Sommerfeld enhancement is therefore significant for $M_V < |\alpha|M$, and at temperatures $T \lesssim \alpha^2 M$. Furthermore, $S$ is resonantly enhanced when $M = \kappa n^2 M_V/\alpha$ for integer $n$, which corresponds to a zero-energy bound state (its decay width mitigates the resonant enhancement, see Blum et al. in [132]).[9]

When Sommerfeld enhancements are relevant, a second related phenomenon is also important: the formation of bound states of two Dark Matter particles with binding energy of order $\alpha^2 M$, analogous to the formation of hydrogen at recombination [133]. Since the two DM particles within the bound state annihilate at a rate $\Gamma_{\rm ann} \sim \alpha^5 M$, which is often cosmologically fast, the bound state effects can be described trough a modified effective cross section, without writing a Boltzmann equation for each bound state, see e.g. Mitridate et al. (2017) in [133].

Finally, these effects become more dramatic is DM is a bound state that forms at a QCD-like confinement scale $\Lambda$ that is much smaller than the DM mass $M$.[10] Depending on the values of $M$, $\Lambda$ and of the consequent dark gauge coupling $\alpha$, the cosmological cross section of eq. (4.13) needs to be equal either to the perturbative cross section $\sim \alpha^2/M^2$ (if freeze-out occurs at $T \lesssim M$), or to the non-perturbative cross section $\sim 1/\Lambda^2$ (if non-perturbative bound states recouple later), or to $1/M^2\alpha^2$ (if perturbative bound states play a key role). As a result the indirect-detection DM annihilation cross section predicted by cosmology can be $(M/\Lambda)^2$ larger than the standard value in eq. (4.13).

## 4.1.6  $2 \leftrightarrow 0$ co-annihilations

DM might be part of a 'dark sector' that contains extra particles $DM'$ in addition to DM. For example, in supersymmetric models DM can be the lightest supersymmetric particle; in composite DM models

---

[9]If the long-range force is a non-abelian gauge interaction with vectors $V^a$, the potential between particles in representations with generators $T_R$ and $T_{R'}$ becomes a matrix in component space. As long as the group is unbroken, its algebra allows to decompose the processes into effectively abelian sub-sectors, $R \otimes R' = \sum_J J$, as

$$V = \alpha\frac{e^{-M_V r}}{r}\sum_a T_R^a \otimes T_{R'}^a = \alpha\frac{e^{-M_V r}}{2r}\left[\sum_J C_J 1_J - C_R 1_R - C_R 1_{R'}\right], \tag{4.31}$$

where $C$ are the quadratic Casimirs ($C_n = (n^2 - 1)/4$ for DM in an $n$ dimensional representation of SU(2); $C_N = (N^2 - 1)/2N$ for DM in a fundamental ($N$-dimensional) representation of SU($N$), etc). In each $J$ sub-sector one gets an effective abelian-like potential

$$V_J = -\alpha_{\rm eff}\frac{e^{-M_V r}}{r}, \qquad \alpha_{\rm eff} = \frac{C_R + C_{R'} - C_J}{2}\alpha. \tag{4.32}$$

[10]Composite DM will be further discussed in section 9.5.

the dark sector contains excited resonances. Extra dark particles with masses $M'$ need to be taken into account in cosmological computations of the relic abundance if $M' - M \lesssim \text{few} \times T_{\text{dec}}$ where $T_{\text{dec}} \sim M/25$ is the decoupling temperature. The thermal abundance of DM$'$ is suppressed relative to the DM one by a Boltzmann factor $e^{(M-M')/T}$ so that for larger mass differences the abundance of DM$'$ becomes too small to be important, reverting to the limit of a simple DM thermal freeze out discussed in section 4.1.2.

In the case of $N$ DM states with similar masses, $m_1 \leq m_2 \leq \cdots \leq m_N$, the Boltzmann equation for the number density $n_i$ for each of the particle species, $i = 1, \ldots, N$, is given by [134, 135]

$$
\begin{aligned}
\frac{dn_i}{dt} = & - 3Hn_i - \sum_{j=1}^{N} \langle \sigma_{ij} v_{ij} \rangle \left( n_i n_j - n_i^{\text{eq}} n_j^{\text{eq}} \right) - \sum_{j \neq i} \left[ \Gamma_{ij}(n_i - n_i^{\text{eq}}) - \Gamma_{ji}(n_j - n_j^{\text{eq}}) \right] \\
& - \sum_{j \neq i} \left[ \langle \sigma_{Xij} v_{iX} \rangle \left( n_i n_X - n_i^{\text{eq}} n_X^{\text{eq}} \right) - \langle \sigma_{Xji} v_{jX} \rangle \left( n_j n_X - n_j^{\text{eq}} n_X^{\text{eq}} \right).
\end{aligned}
\tag{4.33}
$$

The first term on the r.h.s. gives the dilution of $n_i$ due to the expansion of the Universe. The second term describes the effects of the DM$_i$ + DM$_j$ *co-annihilations*, with the annihilation cross sections $\sigma_{ij} = \sum_X \sigma(\text{DM}_i + \text{DM}_j \to X)$, and the relative velocities in the thermal average given by $v_{ij} = \sqrt{(p_i \cdot p_j)^2 - m_i^2 m_j^2}/E_i E_j$. Note that in this notation DM$_1$ is the stable DM particle. The third term in eq. (4.33) gives the contributions from the DM$_i$ decays, with decay widths given by $\Gamma_{ij} = \sum_X \Gamma(\text{DM}_i \to \text{DM}_j + X)$. The terms in the second line of (4.33) describe the DM$_i \to$ DM$_j$ conversions on the SM thermal background (or on any other additional particles in the cosmic thermal plasma), with cross sections $\sigma_{Xij} = \sum_{X'} \sigma(\text{DM}_i + X \to \text{DM}_j + X')$.

While the coupled Boltzmann equations for $n_i$ in eq. (4.33) can be complicated, the problem simplifies significantly if we are only interested in the relic abundance of DM. The relic number density of DM is given by $n = \sum_i n_i$, as long as all the other states eventually decay to DM (plus SM states). (If this is not the case we are in a regime of *multi-component DM*.) The Boltzmann equation governing the time evolution of $n$ is much simpler,

$$
\frac{dn}{dt} = -3Hn - \sum_{i,j=1}^{N} \langle \sigma_{ij} v_{ij} \rangle \left( n_i n_j - n_i^{\text{eq}} n_j^{\text{eq}} \right),
\tag{4.34}
$$

since in the sum the third and the fourth terms in eq. (4.33) separately cancel. That is, for the total number of DM particles the conversions DM$_i \leftrightarrow$ DM$_j$ do not matter because they conserve the total DM number.

Further simplification is possible since the background number densities $n_X$ are much larger than the already Boltzmann suppressed DM number densities $n_i$, while the annihilations cross sections $\sigma_{ij}$ and $\sigma_{iX}, \sigma_{j,X}$ are usually comparable. This means that the last term in eq. (4.33) ensures that DM$_i$ remain in thermal equilibrium, and in particular $n_i/n \simeq n_i^{\text{eq}}/n_{\text{eq}}$. This then gives

$$
\frac{dn}{dt} = -3Hn - \sum_{i,j=1}^{N} \langle \sigma_{\text{eff}} v \rangle \left( n^2 - n_{\text{eq}}^2 \right),
\tag{4.35}
$$

where the effective annihilation cross section is given by

$$
\langle \sigma_{\text{eff}} v \rangle = \sum_{ij} \langle \sigma_{ij} v_{ij} \rangle \frac{n_i^{\text{eq}}}{n_{\text{eq}}} \frac{n_j^{\text{eq}}}{n_{\text{eq}}}.
\tag{4.36}
$$

From the expression for $\langle \sigma_{\text{eff}} v \rangle$ we see that in the case where there are large mass splittings between DM$_i$ and DM$_1$, the contributions from DM$_i$ are exponentially suppressed due to the Boltzmann suppres-

sions, $n_i^{\mathrm{eq}} \ll n_1^{\mathrm{eq}}$ and thus $n_{\mathrm{eq}} \simeq n_1^{\mathrm{eq}}$. In this limit we recover the expression for the single DM thermal freeze out. For $m_i \simeq m_1$, on the other hand, all the $\mathrm{DM}_i$ contribute to the effective annihilation cross section. In particular, the $\mathrm{DM}_1 + \mathrm{DM}_1$ annihilation can be heavily suppressed, and nevertheless $\langle \sigma_{\mathrm{eff}} v \rangle$ can be large due to the contributions from $\mathrm{DM}_i$, $i \neq 1$. This is the case, for instance, when $\mathrm{DM}_i$ carry color charges and thus have large annihilation cross sections controlled by QCD, typically much larger than the $\mathrm{DM}_1 + \mathrm{DM}_1$ annihilation cross section. Without such additional annihilations the $\mathrm{DM}_1$ would have had a much larger relic abundance $\Omega_{\mathrm{DM}} \propto 1/\langle \sigma_{11} v_{11} \rangle$. However, due to the presence of extra $\mathrm{DM}_i$ states the excess $\mathrm{DM}_1$ convert to $\mathrm{DM}_i$ through scatterings on the SM background, and then the $\mathrm{DM}_i$ annihilate away, resulting in a smaller relic abundance than if $\mathrm{DM}_i$ were not present (that is, in this case $\langle \sigma_{\mathrm{eff}} v \rangle$ is larger than $\langle \sigma_{11} v_{11} \rangle$).

The converse can also be true. If $\sigma_{ij} \ll \sigma_{11}$, it is possible that $\langle \sigma_{\mathrm{eff}} v \rangle$ is smaller than $\langle \sigma_{11} v_{11} \rangle$. In this situation $\mathrm{DM}_1$ annihilates away through $\mathrm{DM}_1 + \mathrm{DM}_1$ process, but is being repopulated through $\mathrm{DM}_i \to \mathrm{DM}_1$ conversions on the cosmic thermal plasma background. In this case the DM relic abundance is larger than if the extra $\mathrm{DM}_i$ states with small annihilation cross sections were not present.

## 4.1.7  $2 \leftrightarrow 1$ semi-annihilations

Recalling that we count only the number of DM-sector particles, so that any number of SM particles can be present, the $2 \leftrightarrow 1$ notation means that we now allow for extra processes such as

$$\mathrm{DM}_i \mathrm{DM}_j \leftrightarrow \mathrm{DM}_k \mathrm{SM}, \tag{4.37}$$

where SM is one or more particles in the SM sector. These processes, known as *semi-annihilation* [136], would add an extra term $-\gamma_{2 \leftrightarrow 1}(Y^2/Y_{\mathrm{eq}}^2 - Y/Y_{\mathrm{eq}})$ on the r.h.s. of the Boltzmann equation (4.15). Their effect is similar to annihilations. Semi-annihilations arise in models where DM particles are stable because of a symmetry different than a $\mathbb{Z}_2$ reflection, $\mathrm{DM} \to -\mathrm{DM}$. A concrete example is DM whose constituents are the massive vectors $Z', W'$ of a dark $\mathrm{SU}(2)'$ gauge group, broken by a dark scalar doublet $H'$ that mixes with the Higgs doublet $H$ [136]. Cubic gauge interactions among dark vectors lead to semi-annihilations. Such DM is stable for the same reason that the SM gauge bosons $Z, W$ would have been stable, had the SM fermions not been present.

In these kinds of models all the DM states can be stable, if the decay processes $\mathrm{DM}_k \to \mathrm{DM}_i \mathrm{DM}_j \mathrm{SM}$ are kinematically forbidden. For example, in the SM without electroweak interactions the proton $p$, the neutron $n$, and the charged pions $\pi^\pm$ would all be stable, despite the presence of semi-annihilations such as $p\pi^- \to n\pi^0$ (where $\pi^0$ would decay into $\gamma\gamma$ as usual).

Note that, if DM initially only has a small abundance, it is increased exponentially by the inverse semi-annihilations, the $1 \to 2$ scatterings [136].

The $2 \leftrightarrow 1$ processes of the form $\mathrm{DM}_i \mathrm{DM}_j \mathrm{SM} \leftrightarrow \mathrm{DM}_k \mathrm{SM}$, which have in the initial state an extra SM particle with low abundance (such as relic protons with well below thermal abundance, $n_p \ll n_\gamma$), behave similarly to the $3 \leftrightarrow 2$ processes to be discussed in section 4.1.10 [137].

## 4.1.8  $1 \leftrightarrow 0$ thermal decays

The notation $1 \leftrightarrow 0$ means that we now consider processes where one DM particle disappears, and any number of SM particles can be present. So, for example, we here study DM undergoing processes of the type

$$\mathrm{DM}\,\mathrm{SM} \leftrightarrow \mathrm{SM}\,\mathrm{SM}. \tag{4.38}$$

These processes, dubbed *thermal decays*, would add an extra term $-\gamma_{1 \leftrightarrow 0}(Y/Y_{\mathrm{eq}} - 1)$ on the r.h.s. of the Boltzmann equation (4.15). If DM scatters on a light SM particle, thermal decays are more efficient than annihilations, because they are less Boltzmann suppressed at temperatures $T \lesssim M$. Consequently, the

final DM abundance can be suppressed by a smaller power of the cross section than in the case where the DM abundance is set purely by annihilations, eq. (4.22).

Thermal decays are only present in theories where there is no $\mathbb{Z}_2$ or any other symmetry that would separate the dark sector from the SM sector, and prevent DM from decaying. In these theories DM therefore undergoes decays also at zero temperature, which is phenomenologically challenging given that DM decays are subject to very strong bounds, as discussed in section 6. The DM decay width must be roughly 30 orders of magnitude smaller than the finite-temperature DM decay rate needed to achieve thermal equilibrium at $T \sim M$, $\gamma_{1\leftrightarrow 0} \approx n_{\mathrm{DM}}\Gamma_{\mathrm{DM}T}$ with $\Gamma_{\mathrm{DM}T} \sim H$. Consequently, the relic DM abundance is set through thermal decays only in very special theories [138].

### 4.1.9   $3 \leftrightarrow 2$ DM processes in a decoupled dark sector

The notation $3 \leftrightarrow 2$ means that we now consider processes where DM-sector particles undergo processes such as

$$\text{DM DM DM} \leftrightarrow \text{DM DM} \tag{4.39}$$

and any number of extra SM particles could be present on both sides. These processes occur when DM has significant self-interactions that lead to number-changing scatterings, for example, through a combination of cubic and quartic interactions of scalar DM. Even though the DM number is no longer conserved, DM can still be stable on cosmological timescales, if in the dark sector there are no other particles lighter than the DM, and the interactions with light SM particles are small enough. An example of a concrete UV model that leads to $3 \leftrightarrow 2$ scattering is DM that is a glue-ball of a new confining non-abelian gauge interaction in the dark sector.

Let us first focus on the simplest case; that of a completely decoupled dark sector [139], without any interactions with the SM. As we will see below, in this case the $3 \leftrightarrow 2$ scatterings do not provide a viable mechanism for DM production. This is quite different from the other DM production mechanisms that we studied so far, all of which would have worked also if only the (appropriate) dark sector were involved.

The challenge for the decoupled dark sectors with $3 \leftrightarrow 2$ scatterings is that when the temperature $T_{\mathrm{DM}}$ in the dark sector becomes smaller than $M$, the scatterings such as DM DM DM $\to$ DM DM heat the dark sector, keeping it at $T_{\mathrm{DM}} \sim M$. This was dubbed '**cannibalism**' [139], because the heating occurs at the price of the DM number: DM keeps warm by eating itself. Cannibalism ends at the decoupling temperature $T_{\mathrm{DM}}^{\mathrm{dec}}$, at which point DM gets so diluted that the cannibalistic scatterings become slower than the rate of the expansion of the Universe, and drop out of thermal equilibrium. The predictions for the cannibalistic phase are easy to compute since the comoving entropy $sa^3$ is conserved separately in each of the two decoupled sectors: SM and DM. In the SM sector this gives the usual $T_{\mathrm{SM}} \propto 1/a$. In the DM sector one has $s_{\mathrm{DM}} \equiv (\rho_{\mathrm{DM}} + p_{\mathrm{DM}})/T_{\mathrm{DM}} \approx M n_{\mathrm{DM}}^{\mathrm{eq}}/T_{\mathrm{DM}}$, where the non-relativistic number density $n_{\mathrm{DM}}^{\mathrm{eq}}$ of eq. (C.22) contains the exponential factor $e^{-M/T_{\mathrm{DM}}}$. In view of this factor, during cannibalism $T_{\mathrm{DM}}$ decreases only logarithmically, $T_{\mathrm{DM}} \simeq M/3\ln(a/a_M)$, where $a_M$ is the scale factor at which $T_{\mathrm{DM}} = M$. The possibility that the cannibalistic phase happens during matter domination is subject to constraints, because a cannibalistic dark sector affects large scale structures differently from ordinary dark matter [139]. After decoupling, the dark sector no longer self-interacts, and starts behaving as ordinary dark matter: $\rho_{\mathrm{DM}}$ and $s_{\mathrm{DM}}$ scale as $1/a^3$. The conserved ratio $\xi \equiv s_{\mathrm{SM}}/s_{\mathrm{DM}}$ is of order one if the SM and DM sectors have been initially in equilibrium at $T \gg M$. The predicted DM abundance is therefore

$$Y_{\mathrm{DM}} \equiv \frac{n_{\mathrm{DM}}}{s} \approx \frac{s_{\mathrm{DM}} T_{\mathrm{DM}}^{\mathrm{dec}}/M}{s_{\mathrm{SM}}}. \tag{4.40}$$

The cosmological DM abundance is reproduced for $Y_{\mathrm{DM}} \approx 0.4\,\mathrm{eV}/M$, see eq. (4.3). The DM decoupling temperature is thereby

$$T_{\mathrm{DM}}^{\mathrm{dec}} \approx 0.4\,\mathrm{eV}\,\xi, \tag{4.41}$$

independently of the DM model. Unless $\xi$ is very large, this is problematic in two ways: such low $T_{\rm DM}^{\rm dec}$ would affect standard cosmology and for instance lead to problems with BBN. It also implies that the DM mass is not much above the eV scale, with too large DM self-interactions. Very large $\xi$ can be obtained if the dark sector is populated through freeze-in.

While the simplest version of cannibalism is excluded as the DM production mechanism, more complicated variants of this scenario are allowed. For example, some heavier unstable dark sector particle DM$'$ could undergo 'cannibalistic' $2 \leftrightarrow 3$ scatterings, where the cooling of SM plasma temperature, $T_{\rm SM}$, compared to dark sector one, $T_{\rm DM}$, is interrupted when DM$'$ decays [139]. Alternatively, DM could remain kinetically coupled to the SM during (part of) the cannibalistic phase, so that the SM sector is reheated possibly up to the dark sector temperature [139, 140]. This possibility is studied in the next section.

## 4.1.10   $3 \leftrightarrow N$ and $4 \leftrightarrow N$ in thermal equilibrium

As in the previous section, we assume that DM undergoes processes such as DM DM DM $\leftrightarrow$ DM DM, denoted as $3 \leftrightarrow 2$. However, this time we also assume that some process keeps DM in thermal equilibrium with the SM sector, so that the two have a common temperature [140]. The thermalising interaction can also give DM DM $\leftrightarrow$ SM SM scatterings: we assume that their effect can be neglected in the Boltzmann equation for the DM abundance $Y$

$$sHZz \frac{dY}{dz} = \gamma_{3 \leftrightarrow 2} \left( \frac{Y^2}{Y_{\rm eq}^2} - \frac{Y^3}{Y_{\rm eq}^3} \right). \tag{4.42}$$

This can be compared with the Boltzmann equation in eq. (4.15) for $2 \leftrightarrow 0$ annihilations. The quick estimate of the DM relic abundance for $2 \to 0$ process, presented in section 4.1.1, can be adapted to the $3 \leftrightarrow 2$ case, as follows. DM freezes-out at $T = T_{\rm dec}$, when the DM interaction rate is comparable to the Hubble rate $\Gamma_{3 \leftrightarrow 2} \sim H$. The interaction rate $\Gamma_{2 \leftrightarrow 0} = \langle \Phi \sigma_{2 \leftrightarrow 0} \rangle$ gets replaced by $\Gamma_{3 \leftrightarrow 2} \approx \langle \Phi^2 \sigma_{3 \leftrightarrow 2} \rangle$. Here $\Phi = n_{\rm DM} v$ is the DM flux, and dimensional analysis fixes the $3 \to 2$ analog of the $2 \to 0$ cross section $\sigma_{2 \leftrightarrow 0}$ to have the form $\sigma_{3 \leftrightarrow 2} \sim \alpha^3 / M^5$, where $g$ is some DM self-coupling and $\alpha = g^2 / 4\pi$. Omitting order one factors, we have $T_{\rm dec} \sim M$ and $v \sim 1$, and the relic DM density is

$$Y \sim \frac{1}{M^2 \sqrt{M_{\rm Pl} \sigma_{3 \leftrightarrow 2}}} \sim \sqrt{\frac{M}{M_{\rm Pl} \alpha^3}}. \tag{4.43}$$

Note that $Y$ is suppressed by a smaller power of $M_{\rm Pl}$ than in the $2 \leftrightarrow 0$ case. As a result the observed DM cosmological abundance, $Y \sim {\rm eV}/M$, is matched for lighter DM

$$M/\alpha \sim ({\rm eV}^2 M_{\rm Pl})^{3/2} \sim {\rm GeV}. \tag{4.44}$$

In addition to $3 \leftrightarrow 2$ DM processes, specific models can predict $2 \leftrightarrow 2$ scatterings among DM, with cross section $\sigma/M \sim \alpha^2/M^3$. This are potentially large enough that they risk conflicting with the bullet cluster bound in eq. (1.5), unless $M$ is as heavy as possible. Eq. (4.44) then implies large $\alpha$.

So far we assumed full thermal equilibrium between DM and the SM, while the opposite was assumed in section 4.1.9. An intermediate regime, where thermalization interactions decouple, while the $3 \leftrightarrow 2$ scatterings are still active, is also possible, see [140] for further details.

The DM models with $3 \leftrightarrow 2$ scatterings that received quite some attention have dark pions as DM states, with some SM interactions (see section 9.5.1). Another possibility discussed in the literature is quartics among dark scalars, imposing $\mathbb{Z}_3$ or $\mathbb{Z}_5$ symmetries. These could possibly be remnants of a dark U(1) gauge symmetry broken by a scalar with charges 3 or 5 [140].

Note that having 2 DM particles in the final state played no role in the estimate in eq. (4.44). Therefore the same estimate, up to order unity factors, also applies to the case of pure $3 \leftrightarrow 0$ or $3 \leftrightarrow 1$ processes (we recall that our notation counts the number of dark sector particles, allowing for any number of extra SM particles). Possible models for such processes can be obtained considering dark scalars with quartic couplings, such as $SSSS'$. However, these $3 \leftrightarrow 0, 1$ processes have little interest because they are accompanied by more efficient $2 \leftrightarrow 1, 2$ scatterings, obtained by moving one dark sector particle from the initial to the final state (these related processes can be suppressed in special models with a large asymmetry in the dark sector).

Finally, we consider scatterings among more the 3 dark-sector particles. The estimate in eq. (4.44) can be extended to the case of $4 \leftrightarrow 2$ DM annihilations: the corresponding rate is given by $\Gamma_{4\leftrightarrow2} \sim \Phi^3 \sigma_{4\leftrightarrow2}$ with $\sigma_{4\leftrightarrow2} \sim \alpha^4/M^8$, leading to the estimate $M/\alpha \sim (\,\mathrm{eV}^3 M_{\mathrm{Pl}})^{1/4} \lesssim \mathrm{MeV}$ for the DM mass. This is typically excluded.

## 4.1.11   Special freeze-out kinematical configurations

Specific limits of the thermal freeze-out mechanism can lead to significant deviations from the generic estimates discussed above, with phenomenological implications. Examples include the following.

1. The '**forbidden DM**' limit occurs when DM is lighter than any of the SM states it can annihilate into [141]. This means that $\mathrm{DM\,DM} \to \mathrm{SM\,SM}$ annihilation is kinematically open for $T \gg M$, while at the freeze-out temperatures only the DM particles on the higher-energy tail of the thermal distribution can annihilate. This scenario can be realized, if DM has negligible couplings to the SM particles that are lighter than DM (such as photons and neutrinos). 'Forbidden DM' evades CMB bounds on light DM, because DM annihilations are highly suppressed at the low temperatures relevant for the CMB. The CMB bounds are evaded even if the splittings between DM and the states it annihilates to are tiny (this scenario was termed the '**impeded DM**' [142]).

2. DM DM annihilations can be mediated by a resonance in the $s$-channel and be **near the pole** for specific DM energies. This was discussed in section 4.1.5.

3. **Decoupling of inelastic scattering** (or the so called **co-scattering**) occurs when DM DM annihilations are negligible, while the DM cosmological abundance decreases through inelastic scatterings, $\mathrm{DM\,SM} \to \mathrm{DM'\,SM}$, followed by $\mathrm{DM'\,DM'}$ annihilations [143]. The required scattering cross section is very different from the simple thermal freeze-out; in particular, the DM mass is generically much lighter than the weak scale. Furthermore, the DM DM annihilation rate can be arbitrarily small, thus avoiding CMB constraints on light DM.

4. **Co-annihilation of split states.** Usually, co-annihilation of DM is important when there are dark states with masses similar to the DM particle. However, if the DM self-annihilation is suppressed, co-annihilations with states in the dark sector significantly heavier than DM become relevant for setting the DM relic abundance. This limit can lead to the correct relic abundance even for very light DM, down to the keV scale [144] (see also section 4.1.6).

## 4.1.12   Special freeze-out cosmological configurations

So far we assumed standard cosmology while DM freezes out, at $T \sim M/25$. This needs not be the case, since the DM freeze out likely occurred before the Big Bang Nucleosynthesis, at temperature well above $T \gtrsim \mathrm{MeV}$ and cosmology is largely untested at such high temperatures. A non-standard cosmological history can result in significant deviations from the generic estimates discussed above, with essentially endless variations still possible. Concrete examples include:

1. If the Universe underwent a phase of fast expansion, this affected the DM relic abundance [145]. For example, if the Universe contains an exotic fluid whose energy density evolved as $\rho \propto a^{-3(1+w)}$ with $w > 1/3$, and thus redshifted faster than radiation (see eq. (C.8)), this fluid dominated the energy budget of the Universe at very early times, before the era of radiation domination. This induced a phase of fast expansion. A concrete realization discussed in the literature is the so called **'kination'** phase, in which the energy budget of the Universe is dominated by the kinetic energy of a time-dependent scalar field. This could be the quintessence field, which in the late stages of the evolution of the Universe gives rise to Dark Energy.

   Since in this case the Universe during freeze-out expands faster than it does in the standard cosmology, the correct DM abundance is reproduced via freeze-out with a DM annihilation cross section $\langle \sigma v \rangle$ *larger* than the standard one of eq. (4.13). Because of the larger annihilation cross-section, DM keeps annihilating longer, after departing from thermal equilibrium. This scenario has thus also been termed '**relentless DM**' [145].

2. The Universe is believed to have gone through a phase of exponential inflationary expansion at very early times, see section 4.3. If the Universe has also undergone a later short period of fast expansion, sometimes called 'late time inflation', then the density of any particle species (including DM) that had already decoupled by the beginning of the late time inflationary period would have been significantly **diluted** [146]. Once late-time inflation ends, its energy density reheats light SM particles while DM is too heavy. In these models, DM needs an annihilation cross section $\langle \sigma v \rangle$ much *smaller* than what is needed for the standard thermal freeze-out, and that would have otherwise lead to an over-abundant DM in the standard cosmology. The idea of inflationary dilution is simple and general and can be applied to other models featuring over-abundant DM, e.g., to axions with 'too large' $f_a$ (see section 10.4.4). A specific realization of this setup is discussed in section 4.5.3.

3. A different kind of dilution of the DM density occurs if, after DM freeze-out, the Universe underwent a phase of **early matter domination** [147]. This happens if the energy budget of the Universe becomes temporarily dominated by metastable particles. If these are sufficiently heavy and their lifetimes long enough (but short enough to avoid conflicts with Big Bang Nucleosynthesis bounds), they may temporarily dominate the energy density of the Universe, so that, when they eventually decay, they **inject significant entropy** in the SM bath and thus dilute any relic abundance, including that of DM [147]. Assuming that the mass of the decaying particle is sufficiently small ($M' < 2M$), guarantees that its decays can not reheat DM itself. If the freeze-out had resulted in overabundant DM, even up to equilibrium-like abundances $n_{\rm DM} \sim n_\gamma$, the mechanism can then dilute it down to the observed density.

The latter two possibilities allow for a smaller cross section at freeze-out time, and therefore for a *larger DM mass*, possibly above the unitarity limit (see section 4.1.4). In concrete models these extra decaying particles can belong to the dark sector, or mediate interactions between the dark and the SM sectors (an example of such a mediator is a $Z'$ vector boson).

## 4.2   DM as a non-thermal relic

Freeze-out from thermal equilibrium is a generic mechanism that applies to any interacting massive particle. One of its main attractive features is that in its simplest form it predicts the DM relic abundance just from annihilation cross section (with very small dependence on DM mass). Many other, less predictive scenarios are also possible. Below, we briefly summarize some of them.[11]

---

[11]For a systematic appraisal of DM production mechanisms, including freeze-out, freeze-in and others, see also Chu et al. (2012) [148].

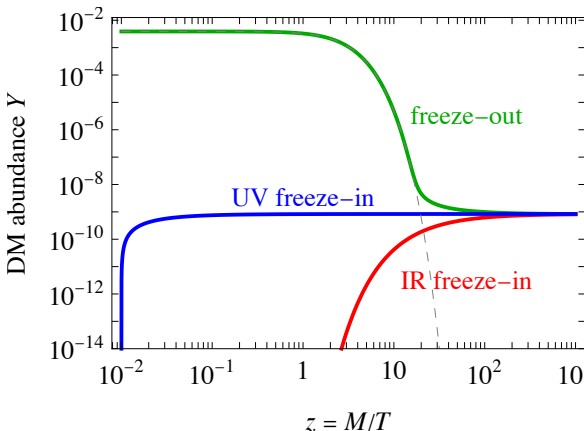

Figure 4.3: *Illustration of the cosmological evolutions of DM abundance $Y = n/s$ for the IR (red curve) and the UV (blue) freeze-in, compared to the freeze-out (green), and the thermal equilibrium abundance (dashed).*

### 4.2.1   Freeze-out and decay

A rather straightforward extension is that the thermal relic is not the DM itself, but rather another particle $X$, which decays into DM after freeze-out [149]. This results in a DM density $\Omega_{\mathrm{DM}} = \Omega_X M/M_X$, up to extra model-dependent corrections, e.g., if the decay is slow enough that the decays reheat the universe. Since $M_X > M$ the correct relic abundance is obtained for smaller annihilation cross sections than eq. (4.13). Gravitino DM (see section 10.1.2) is a good example of a realization of this scenario.

### 4.2.2   Freeze-in

A possible explanation for the small cosmological density of DM, $Y_{\mathrm{DM}} \approx 0.4\,\mathrm{eV}/M \ll 1$ for $M \gg$ eV, could be that DM is a particle that is so weakly coupled to the rest of the Universe that its initial abundance was vanishingly small. Small interactions of DM with the SM thermal bath then produced a sub-thermal DM abundance $Y_{\mathrm{DM}} \ll 1$, given by

$$Y_\infty = \int_0^\infty \frac{dz}{z} \frac{N\gamma}{Hs} \sim \max_T \frac{N\gamma}{Hs}, \tag{4.45}$$

where $\gamma$ is the space-time density of SM interactions that produce $N$ (usually 2) DM particles. In this situation, the primordial DM velocity distribution would contain information about the production mechanism. This can happen in two main qualitatively different ways, illustrated in fig. 4.3:

▼ **IR freeze-in**. DM particles which have *small renormalizable interactions g* with the SM are dominantly produced at the lowest possible temperatures, typically at temperatures that are a factor of a few below their mass $M$ [150].[12] The space-time density of scatterings that produce DM particles from the SM plasma is roughly $\gamma \sim g^4 T^4$. The resulting DM abundance is $Y_{\mathrm{DM}} \approx \gamma/Hs|_{T \sim M} \sim g^4 M_{\mathrm{Pl}}/M$, which matches the cosmological DM density in eq. (4.3) for $g \sim (Y_{\mathrm{DM0}} M/M_{\mathrm{Pl}})^{1/4} \sim (T_0/M_{\mathrm{Pl}})^{1/4} \sim 10^{-8}$. For example, fig. 3.5 shows that DM with a small electric charge needs $qe \approx 10^{-11}$ for any DM mass $M \lesssim T_{\mathrm{RH}}$ (the reheating temperature $T_{\mathrm{RH}}$ is the

---

[12]A variation is the 'Leak-in DM' scenario [151], in which the small interactions do not produce directly the DM particles but rather populate the hidden sector which subsequently thermalizes and therefore generates DM particles internally.

largest temperature achieved in the evolution of the Universe and thus sets the upper bound on the mass of the heaviest particle that can be produced). Since the freeze-in DM is predominantly produced in the mildly relativistic regime, the precise computations depend on the details about DM interactions: the non-relativistic cross section $\sigma_0$ that we introduced when we were studying the freeze-out, see eq. (4.17), can only be used to obtain an indicative value, $Y_{\text{DM}} \approx g_{\text{SM}} M M_{\text{Pl}} \sigma_0 / g_{\text{SM}}^{3/2}$. A precise computation is possible, if DM is dominantly produced via *decays* of a heavier particle into $N$ DM particles (the heavy particle has mass $M'$, $g'$ degrees of freedom, and decay width $\Gamma'$). The DM production rate is given by eq. (C.36), resulting in

$$Y_{\text{DM}} = \int_0^\infty \frac{dz}{z} \frac{\gamma_{\text{decay}}^{\text{eq}}}{Hs} = \frac{405 N \sqrt{5} M_{\text{Pl}} g' \Gamma'}{16\pi^{9/2} g_{\text{SM}}^{3/2} M'^2}. \tag{4.46}$$

Unlike freeze-out, the IR freeze-in allows for DM to be lighter than an MeV, without conflicting with the big-bang nucleosynthesis constraints.

▲ **UV freeze-in**. DM particles which possess *small non-renormalizable interactions* with the SM particles are dominantly produced at the highest possible temperature $T_{\text{RH}}$ and thereby their abundance depends on $T_{\text{RH}}$. Since the value of $T_{\text{RH}}$ is unknown, there are no unambiguous predictions. The reheating temperature after inflation with Hubble constant $H_{\text{infl}}$ is $T_{\text{RH}} \sim \sqrt{M_{\text{Pl}} H_{\text{infl}}}$, if the inflaton $\phi$ decays promptly, and is lower otherwise, given by $T_{\text{RH}} \sim \sqrt{M_{\text{Pl}} \Gamma_\phi}$, where $\Gamma_\phi$ is the decay width of the inflaton. Indeed, in the latter case, the entropy release from inflaton decay ends when $\Gamma_\phi \sim H \sim \sqrt{\rho_\phi}/M_{\text{Pl}}$ giving $T_{\text{RH}}^4 \sim \rho_\phi$. UV freeze-in can work with larger couplings if the DM mass is mildly above $T_{\text{RH}}$. If the non-renormalizable operators are mediated by renormalizable interactions of particles lighter than $T_{\text{RH}}$, then one reverts back to the IR freeze-in: the DM particles are predominantly produced at temperatures close to the mass of the mediator.

Gravity is one known non-renormalizable interaction, roughly corresponding to $g \sim T/M_{\text{Pl}}$. Gravitational DM production can be estimated as follows:

- Purely gravitational scatterings give $\gamma \sim T^8 e^{-2M/T}/M_{\text{Pl}}^4$ resulting in $Y_{\text{DM}} \approx \gamma/Hs|_{T=T_{\text{RH}}} \sim (T_{\text{RH}}/M_{\text{Pl}})^3 e^{-2M/T_{\text{RH}}}$. DM particles with only gravitational interactions are discussed in section 9.8.

- A special case is the gravitino (see section 10.1.2), a spin $3/2$ fermion motivated by supersymmetry, whose dominant interactions to the SM, such as the coupling to gluons and gluinos, can be of gravitational strength. Gravitinos can then be produced in processes that combine QCD with gravitational interactions, resulting in $\gamma \sim g_s^2 T^6 e^{-M/T}/M_{\text{Pl}}^2$. Inserting $g_s \sim 1$ and assuming $M \ll T_{\text{RH}}$ (such that the gravitino is long-lived), the gravitino number abundance is $Y_{\text{DM}} \approx \gamma/Hs|_{T=T_{\text{RH}}} \sim T_{\text{RH}}/M_{\text{Pl}}$, corresponding to $\Omega_{\text{DM}} \sim (M/\,\text{TeV})(T_{\text{RH}}/10^{10}\,\text{GeV})$ [152]. Another example is KK gravitons, that will be discussed in section 10.3.

It is worth mentioning that the dominant DM interaction being $1 \to N$ decay with rate $\Gamma$, rather than a $2 \leftrightarrow 2$ scattering, is phenomenologically not viable. The decays would give a freeze-in abundance $Y_{\text{DM}} \sim \Gamma/H$ at $T \sim M$, so $Y_{\text{DM}} \sim M_{\text{Pl}}\Gamma/M^2$. The resulting DM abundance $\Omega_{\text{DM}} \sim M_{\text{Pl}}\Gamma/M\,\text{eV}$ is negligible, $\Omega_{\text{DM}} \sim 10^{-8}$, even in the most favorable case: $M \sim$ keV and $\Gamma \sim 1/(10^{10}\,\text{yr})$. Experimental bounds on realistic models (see e.g. 9.2.2) give much stronger constraints on $\Gamma$.

Before accepting freeze-in as a viable possibility, one must consider whether this mechanism generates DM inhomogeneities compatible with observations. Importantly, perturbations on (current) cosmological scales need to be adiabatic rather than iso-curvature. Adiabatic means that there is a unique fluid with the same SM/DM relative density everywhere, while iso-curvature means that there are entropy fluctuations in the relative density. In the case of freeze-out, adiabaticity is enforced by thermal equilibrium [125].

In the case of freeze-in, adiabaticity holds on scales larger than the (small) horizon at freeze-in, because different regions of the universe can be seen as 'separated universes' that undergo the same freeze-in dynamics up to the time delay $\delta t(t, \boldsymbol{x})$. Since the time delay is common to all components in a particular patch of the universe, the result are adiabatic inhomogeneities [125].

## 4.3   DM production from inflation

Many of the cosmological observations indicate that the Universe underwent a phase of cosmological inflation, during which the energy budget was dominated by the potential energy of an 'inflaton' scalar field $\phi$ [21]. A stage of inflation in the early part of the cosmological evolution can explain the observed flatness and the almost uniform energy density common to regions that came into causal contact only during the late cosmological times. Furthermore, quantum fluctuations during inflation provide a source for the small primordial inhomogeneities seen in the CMB and in the matter distribution. In typical models the inflaton potential $V(\phi)$ is quasi-flat away from its minimum, while around the minimum $V(\phi) \simeq m_\phi^2 \phi^2/2$ (the minimum was conveniently chosen to be at $\phi = 0$). The inflaton energy density $\rho_\phi = \dot{\phi}^2/2 + V \simeq V$ results in an inflationary phase, during which the scale factor of the Universe expands as $a \propto \exp(H_{\mathrm{infl}} t)$, with the inflationary Hubble constant given by

$$H_{\mathrm{infl}} = (8\pi V/3 M_{\mathrm{Pl}}^2)^{1/2}. \tag{4.47}$$

After the end of the inflationary period the inflaton energy is converted into particles, resulting in a thermal bath characterized by the reheating temperature $T_{\mathrm{RH}}$.

In some models of inflation the inflaton $\phi$ is identified with the Higgs boson. In general, however, the particle-physics properties of $\phi$, such as its mass $m_\phi$ and the couplings to the SM and DM particles, are unknown. Given that the nature of DM is also unknown, it is conceivable that DM could be produced during inflation. This could occur in many different ways, since there are so many unknowns.[13] Below, we limit the discussion to the minimal mechanisms for DM production:

a) DM production from quantum fluctuations during inflation. This is discussed in sections 4.3.2 (for the case of heavy fields) and 4.3.3 (for the case of light scalars).

b) After inflation ends, the inflaton oscillates around the minimum of its potential. In the simplest case this induces the particle physics process $\phi\phi \to \mathrm{DM\,DM}$, which is kinematically allowed for $m_\phi \gtrsim M$. Enhanced production is obtained in more complicated situations, e.g., if the DM mass $M$ depends on $\phi$ so that DM becomes light around some special value $\phi_*$.

c) When the inflaton decays, this reheats the SM particle bath, and possibly generates a DM abundance. The phase space for $\phi \to \mathrm{DM\,DM}$ decays is open, if $m_\phi \gtrsim 2M$. This is discussed in section 4.3.1.

### 4.3.1   Inflaton decay to DM

When the inflaton decays, it can produce pairs of SM particles as well as pairs of DM particles, as long as $m_\phi \gtrsim 2M$, as mentioned above. Two distinct cases can be considered.

If DM couples strongly enough to the SM, the two sectors thermalise, i.e., they acquire a common temperature that is independent of the inflaton couplings (apart from being strong enough). DM can then,

---

[13]The inflaton itself can be the DM, but in a trivial way (see section 10.2.2). If the inflaton action is symmetric under $\phi \to -\phi$, the coherent inflaton field fragments into quanta that become stable DM candidates when inflation ends at $\phi = 0$. If the inflaton couples sufficiently strongly to the SM, the system thermalises and inflaton DM can later undergo thermal freeze-out, reducing to the situation discussed in section 4.1.

during the later stages of the cosmological evolution, undergo a freeze-out, as discussed in section 4.1, and acquire the relic abundance that matches observations.

The other possibility is that DM is so weakly coupled to the SM that it does not thermalise with the SM bath. If DM is also weakly coupled to the inflaton, a small branching ratio for the inflaton decays into DM can then suffice to give rise to the observed cosmological DM abundance:

$$\Omega_{\rm DM} \equiv \frac{\rho_{\rm DM}}{\rho_{\rm cr}} = \frac{s_0 M}{3H_0^2/8\pi G_N} {\rm BR}({\rm inflaton} \to {\rm DM}) \approx \frac{0.110}{h^2} \times \frac{M}{0.40\,{\rm eV}}\,{\rm BR}({\rm inflaton} \to {\rm DM}). \tag{4.48}$$

In the numerical evaluation above we used the present entropy density $s_0 = g_{s0}T_0^3 2\pi^2/45$ with $g_{s0} = 43/11$, the present Hubble constant $H_0 = h \times 100\,{\rm km/sec}\,{\rm Mpc}$, and the present temperature $T_0 = 2.725\,{\rm K}$. The observed DM abundance is thereby reproduced for ${\rm BR}({\rm inflaton} \to {\rm DM}) \approx 0.40\,{\rm eV}/M$. In this context, DM could be a particle with undetectably small couplings to the SM. The needed small branching ratio of the inflaton into the dark sector can be realized relatively easily: the SM features a light Higgs scalar, which can have a large (super)renormalizable coupling to the inflaton, while it is conceivable that the dark sector particles couple to the inflaton only through suppressed operators. This happens, for example, if the gravitational action contains an $R^2$ curvature term (equivalent to having an additional scalar with only gravitational couplings) as well as a $\xi|H|^2 R$ term [153].

## 4.3.2 Quantum fluctuations during inflation: heavy fields

If DM does not directly couple to the inflaton, the only DM production taking place during inflation is from quantum fluctuations. This production is always present given that gravity already leads to a contribution [154]. It is interesting to discuss this effect, even though it is in general sub-leading, since it is less model-dependent than the other two production mechanisms, from inflaton oscillations and from inflaton decays.

A simple analogy illustrates how an expanding Universe leads to particle production: a harmonic oscillator in its ground state is described by the wave function $\psi(x) \propto e^{-m\omega x^2/2\hbar}$. If the spring constant $m\omega$ is varied suddenly (for the ease of argument) the wave function remains the same, but this means that it now contains excited components when expressed in terms of the new eigenstates. The expansion of the Universe leads to a time dependence of the proper frequencies of free particles, giving rise to a similar phenomenon. The results are qualitatively different for particles that are heavier or lighter than the inflationary Hubble scale. We start with the case of heavy fields, while the case of light fields will be discussed in section 4.3.3.

To calculate the inflationary particle production [155], it is useful to use the conformal time defined by $d\eta = dt/a$. In this way we can see explicitly that the metric describing a generic homogenous and expanding universe is conformally flat

$$ds^2 = g_{\mu\nu}dx^\mu dx^\nu = dt^2 - a(t)^2 d\boldsymbol{x}^2 = a^2(\eta)[d\eta^2 - d\boldsymbol{x}^2]. \tag{4.49}$$

That is, the metric is given by the Minkowski metric times the scale factor $a$. During inflation, $\eta = -1/aH_{\rm infl} \le 0$.

Fields experience the expansion of the Universe, if their actions contain terms that break conformal symmetry, for example, the mass terms. A non trivial feature of classical massless vectors $V_\mu$ and massless fermions $\psi$ is that their actions are invariant under the Weyl transformation[14] (a massless scalar $\varphi$ respects this symmetry, if its coupling to gravity, $-\xi\varphi^2 R/2$, takes the 'conformal' value $\xi = -1/6$)

$$g_{\mu\nu}(x) \to e^{2\sigma(x)}g_{\mu\nu}(x), \quad \varphi(x) \to e^{-\sigma(x)}\varphi(x), \quad \psi(x) \to e^{-3\sigma(x)/2}\psi(x), \qquad V_\mu \to V_\mu, \tag{4.50}$$

---

[14]Weyl transformation is a combination of a local dilatation and a diffeomorphism, such that the coordinates

where $\sigma$ is an arbitrary function. One can choose $e^{2\sigma} = 1/a^2$, which absorbs the expansion of the Universe, showing that such massless fields are not going to be produced simply due to the expansion of the Universe. The symmetry is, however, broken by nonzero masses, by $\xi \neq -1/6$ (for scalars), and by quantum RG running. For simplicity we focus on the case of a real massive scalar field $\varphi$, though the computation proceeds in a similar way for fermions or vectors.[15] The tree-level scalar action is

$$S = \int d^4x \sqrt{|\det g|} \left[ g^{\mu\nu} \frac{(\partial_\mu \varphi)(\partial_\nu \varphi)}{2} - M^2 \frac{\varphi^2}{2} - \frac{1}{2}(\bar{M}_{\rm Pl}^2 + \xi\varphi^2)R \right]. \tag{4.51}$$

By performing a Weyl transformation, the action for the Weyl-rescaled scalar $\chi \equiv a\varphi$ simplifies such that $a$ only appears in the terms that break conformal symmetry, the mass $M$ and $\xi + 1/6$:

$$S_\chi = \int d^3x\, d\eta \left[ \frac{(\partial^\mu \chi)(\partial_\mu \chi)}{2} - M_{\rm eff}^2 \frac{\chi^2}{2} \right], \qquad M_{\rm eff}^2 = a^2 M^2 + a^2 \left( \frac{1}{6} + \xi \right) R. \tag{4.52}$$

Expanding the field in Fourier modes

$$\chi(\boldsymbol{x}, \eta) = \int \frac{d^3k}{(2\pi)^3} \left[ a_{\boldsymbol{k}} \chi_{\boldsymbol{k}}(\eta) + a_{-\boldsymbol{k}} \chi_{\boldsymbol{k}}^*(\eta) \right] \exp(i\boldsymbol{k} \cdot \boldsymbol{x}), \qquad \chi_{\boldsymbol{k}} = \chi_{-\boldsymbol{k}}, \tag{4.53}$$

the classical equation of motion for the mode $\chi_k$ is given by

$$\frac{d^2 \chi_k}{d\eta^2} + \omega_k^2 \chi_k = 0, \qquad \text{where} \qquad \omega_k^2 = k^2 + M_{\rm eff}^2. \tag{4.54}$$

During de Sitter inflation $a = -1/\eta H_{\rm infl}$ and $R = -12 H_{\rm infl}^2$. Setting $k = 0$ gives the evolution of the homogeneous scalar background, where the solution is a linear combination of two functions (hence the $\pm$ sign in the following)

$$\chi_0 \propto \eta^{\frac{1}{2} \pm \nu} \qquad \Rightarrow \qquad \varphi_0 \propto e^{-(\frac{3}{2} \pm \nu) H_{\rm infl} t}, \qquad \nu^2 = \frac{1}{4} - \frac{M^2}{H_{\rm infl}^2} + 12 \left( \frac{1}{6} + \xi \right). \tag{4.55}$$

A conformally coupled massless scalar corresponds to $\nu^2 = 1/4$. The zero mode grows with time $t$ if $\nu > 3/2$ (one of its components, while the other dies off), decays exponentially if $0 \leq \nu < 3/2$, and oscillates if $\nu^2 < 0$. If $\xi = 0$ the range $0 < \nu < 3/2$ corresponds to $\frac{3}{2} H_{\rm infl} > M > 0$; if $M = 0$ the range corresponds to $-3/16 < \xi < 0$, which includes the conformal value $\xi = -1/6$.

For $k \neq 0$, the dependence of $\omega_k$ on $\eta$ leads to $\chi_k \not\propto e^{-i\omega_k \eta}$. This means particle production, since a generic function can be expanded in the basis of quanta of a given frequency. We assume that during early inflation each mode is initially in its ground state, corresponding to the 'Bunch-Davies' boundary

---

remain unchanged:

| | | dilatation | $\otimes$ | diffeomorphism | $=$ | Weyl transformation |
|---|---|---|---|---|---|---|
| Coordinates | $dx^\mu$ | $e^\sigma dx^\mu$ | | $e^{-\sigma} dx^\mu$ | | $dx^\mu$ |
| Spin 0, scalars | $\varphi$ | $e^{-\sigma}\varphi$ | | $\varphi$ | | $e^{-\sigma}\varphi$ |
| Spin 1/2, fermions | $\psi$ | $e^{-3\sigma/2}\psi$ | | $\psi$ | | $e^{-3\sigma/2}\psi$ |
| Spin 1, vectors | $V_\mu$ | $e^{-\sigma}V_\mu$ | | $e^\sigma V_\mu$ | | $V_\mu$ |
| Spin 2, graviton | $g_{\mu\nu}$ | $g_{\mu\nu}$ | | $e^{2\sigma}g_{\mu\nu}$ | | $e^{2\sigma}g_{\mu\nu}$ |

[15]An abelian vector with (possibly problematic) Stueckelberg mass behaves differently: the production of its longitudinal component is enhanced [156]. On small scales this state behaves like a minimally coupled scalar, while on large scales its production rate is suppressed, thereby avoiding the iso-curvature bounds.

condition $\chi_k(\eta) \simeq e^{-ik\eta}/\sqrt{2k}$ $\eta \to -\infty$. The equation of motion is then solved by

$$\chi_k(\eta) = e^{i(2\nu+1)\pi/4}\sqrt{-\pi\eta}H_\nu^{(1)}(-k\eta), \tag{4.56}$$

where $H_\nu^{(1)} = J_\nu + iY_\nu$ is the Hankel function, with $J_\nu, Y_\nu$ the Bessel functions of the first and second kind, respectively. Only in the conformal case $\nu = 1/2$ the solution reduces to $\chi_k \propto e^{-i\omega_k\eta}$. Performing the rescaling with $a$, this then gives the final state of the scalar field, $\varphi_k$, allowing to compute quantities of interest such as the power spectrum of $\varphi$, its renormalized energy density, etc. Depending on $\nu$, different qualitative behaviours are encountered (we omit the lengthy details of the calculation).

◁ For $\nu^2 > 1/4$ (which, loosely speaking, corresponds to a scalar much lighter than $H_{\mathrm{infl}}$) the scalar $\varphi$ develops large homogenous fluctuations, and acquires a coherent vacuum expectation value $\langle\varphi\rangle$. This results in the 'misalignment' mechanism discussed in section 4.3.4.

▷ For $\nu^2 < 1/4$ the scalar acquires particle fluctuations around $\langle\varphi\rangle = 0$. Loosely speaking, such a 'heavy' scalar acquires a roughy thermal distribution with temperature $T = H_{\mathrm{infl}}/2\pi$, so that, omitting order one factors, the number and energy density at inflation are

$$n_{\mathrm{infl}} \sim H_{\mathrm{infl}}^3 e^{-M/H_{\mathrm{infl}}}, \qquad \rho_{\mathrm{infl}} \approx M n_{\mathrm{infl}}. \tag{4.57}$$

For more general forms of $\omega_k$ (e.g., in the after-inflationary period) there is no analytic solution, and the following approximation scheme is useful. One can parameterize the solution by factoring out the fast free-like oscillations $v_k$,

$$\chi_k(\eta) = \alpha_k(t)v_k(\eta) + \beta_k(\eta)v_k^*(\eta), \qquad v_k(\eta) = \frac{e^{-i\Phi_k}}{\sqrt{2\omega_k}}, \qquad \Phi_k = \int_{\eta_i}^\eta \omega_k(\eta')\,d\eta'. \tag{4.58}$$

The Bogoliubov coefficients $\alpha_k$ and $\beta_k$ are slowly varying, and thus one can neglect $\alpha''$ and $\beta''$ when computing $\chi''$ in the equation of motion, eq. (4.54). The 2nd order equation then simplifies into two 1st order equations

$$\beta_k' = \alpha_k \frac{\omega_k'}{2\omega_k}e^{-2i\Phi_k}, \qquad \alpha_k' = \beta_k \frac{\omega_k'}{2\omega_k}e^{2i\Phi_k}. \tag{4.59}$$

The approximate equation for $\beta_k$ follows immediately, if we further assume that the particles mostly retain their ground-state abundance, $\alpha_k = 1$, giving

$$\beta_k \approx \int_{\eta_i}^\eta \frac{\omega_k'}{2\omega_k}e^{-2i\Phi_k}\,d\eta, \qquad a^3 n = \int \frac{d^3k}{(2\pi)^3}|\beta_k|^2, \tag{4.60}$$

where $n$ is the number density of the produced particles. To complete the computation, one needs to perform lengthy integrals. The results depend on the inflaton model: for instance, during reheating the inflaton can oscillate around its minimum, producing DM from the oscillations (mechanism b) in our list). The expressions (4.60) also allow us to understand the exponential suppression in eq. (4.57) for $M \gg H_{\mathrm{infl}}$: it arises because the Fourier-like integral in $\beta_k$ is suppressed, if the exponential $\exp(-2i\Phi_k)$ oscillates much faster than the cosmological function in front of it.

### 4.3.3  Quantum fluctuations during inflation: light scalar

As mentioned around eq. (4.55), a scalar lighter than the Hubble scale during inflation ($M < \frac{3}{2}H_{\mathrm{infl}}$ for $\xi = 0$) fluctuates, and develops a random vacuum expectation value. The dynamics of coarse-grained slow fluctuations is approximated by the Langevin equation, where the 2nd derivative is neglected and

the effects of fast fluctuations are approximated as a Gaussian noise term $\eta$ (this is a non trivial result, derived in [157]):

$$\frac{d\varphi}{dt} + \frac{1}{3H_{\text{infl}}} \frac{dV}{d\varphi} = \eta(t), \qquad \langle \eta(t)\eta(t') \rangle = \frac{H_{\text{infl}}^3}{4\pi^2} \delta(t - t'). \tag{4.61}$$

This roughly means that there is a fluctuation $\delta\varphi \sim H_{\text{infl}}/2\pi$ per $e$-fold. More precisely, the probability $P(\varphi, N)$ of finding the field at the value $\varphi$ after $N$ $e$-folds obeys the Fokker-Planck equation

$$\frac{\partial P}{\partial N} = \frac{\partial^2}{\partial\varphi^2}\left(\frac{H_{\text{infl}}^2}{8\pi^2} P\right) + \frac{\partial}{\partial\varphi}\left(\frac{V'}{3H_{\text{infl}}^2} P\right). \tag{4.62}$$

Let us discuss solutions for a number of representative cases:

1) A nearly-massless scalar with negligible $V$, and initially at $\varphi = 0$, undergoes a free random walk. After $N$ $e$-folds it acquires a Gaussian distribution with zero mean and a variance

$$\varphi_* = \sqrt{\langle\varphi^2\rangle} = \frac{H_{\text{infl}}}{2\pi}\sqrt{N}. \tag{4.63}$$

2) For larger $M$ the scalar cannot fluctuate freely during inflation. Assuming $V = M^2\varphi^2/2$, the classical motion tends to bring $\varphi$ back to the origin, $\varphi = 0$, in $N \sim (H_{\text{infl}}/M)^2$ $e$-folds. Eq. (4.62) implies in this case that any initial probability converges towards a Gaussian equilibrium distribution with zero mean and variance

$$\langle\varphi^2\rangle = 3H_{\text{infl}}^4/8\pi^2 M^2. \tag{4.64}$$

3) In general, eq. (4.62) implies that after many $e$-folds the probability converges toward the Hawking-Moss distribution

$$P \propto \exp(-8\pi^2 V/3H_{\text{infl}}^4). \tag{4.65}$$

For example, a field with $V = \lambda\varphi^4/4$ converges toward $\langle\varphi^2\rangle = 0.13 H_{\text{infl}}^2/\sqrt{\lambda}$.

The energy stored in these field values can later become DM, as discussed next in section 4.3.4.

## 4.3.4   Initial misalignement

Since $N \approx 60$ $e$-folds of inflation are needed observationally (although many more could have occurred), this means that a very light scalar field $\varphi$ will acquire, through quantum fluctuations, a roughly constant and unknown vacuum expectation value $\varphi_*$, of typical size given in (4.63), generically away from the minimum of the potential at $\varphi = 0$.

We now show that this *initial misalignment* can produce non-relativistic DM with large occupation numbers. The classical equation of motion for a homogeneous DM scalar field after the end of inflation is

$$\ddot{\varphi} + 3H\dot{\varphi} + V'(\varphi) = 0, \tag{4.66}$$

where $H = \dot{a}/a$ is the Hubble expansion rate, while the derivative of the potential, $V'(\varphi)$, can be written as $M^2(\varphi)\varphi$, introducing the field dependent mass $M(\varphi)$ that in general is not merely a constant (this complication will occur in section 10.4, where $\varphi$ will be identified with the axion $a$). The solution to eq. (4.66) has two regimes, depending on the value of $H \sim T^2/M_{\text{Pl}}$:

◁ At early times $t \ll t_*$ the temperature is high and thus $H \gg M$. Both $V'(\varphi)$ and the $\ddot{\varphi}$ term in eq. (4.66) can be neglected, and the vacuum expectation value of $\varphi$ remains frozen at its inflationary value. The field behaves as a vacuum energy.

▷ At later times $t \gg t_*$ one has $H \ll M$: the scalar experiences oscillations that are fast compared to the cosmological expansion rate. The $3H\dot{\varphi}$ term induces a slow change in the oscillation amplitude $A$. Eq. (4.66) can be solved by writing the scalar field as $\varphi(t) = A(t) \cos[\int_{t_*}^t M(\varphi)\, dt]$ and neglecting $A''$ in the equation of motion (the WKB approximation), obtaining

$$A \approx \varphi_* \sqrt{\frac{M(\varphi_*)a_*^3}{M(\varphi)a^3}}. \tag{4.67}$$

This implies that the scalar energy density dilutes in the same way as the non-relativistic matter, $\rho_\varphi \propto |A|^2 \propto 1/a^3$, as we show below.

Averaging over fast oscillations, the energy density and the pressure of the field are given by

$$\rho_\varphi = \left\langle \frac{\dot{\varphi}^2}{2} + \frac{M^2}{2}\varphi^2 \right\rangle = \frac{M^2}{2}A^2, \qquad \wp_\varphi = \left\langle \frac{\dot{\varphi}^2}{2} - \frac{M^2}{2}\varphi^2 \right\rangle = \frac{\dot{A}^2}{2} \ll \rho_\varphi, \tag{4.68}$$

such that $w = \wp_\varphi/\rho_\varphi \approx 0$ and the field behaves as dark matter at $T \ll T_*$. Indeed, multiplying eq. (4.66) by $\dot{\varphi}$ gives $d(\dot{\varphi}^2/2 + V)/dt = -3H\dot{\varphi}^2$ which, on average, becomes the equation of state of matter: $\dot{\rho}_\varphi = -3H\rho_\varphi$. Intuitively, a homogeneous scalar field behaves as non-relativistic matter because it describes a condensate of massive scalars with zero momenta. Their occupation number can be either large (giving the classical field regime) or small (giving the particle regime).

This gives an acceptable cosmology as long as the scalar starts behaving as DM well before the matter/radiation equality at $T_{eq} \sim$ eV, i.e.,

$$M \gg H(T_{eq}) \sim \text{eV}^2/M_{Pl} \sim 10^{-28}\,\text{eV}. \tag{4.69}$$

The induced DM abundance is given in terms of the unknown initial value $\varphi_*$ of the field as

$$\Omega_{DM} = \frac{\rho_\varphi}{\rho_{cr}} \sim \frac{M\varphi_*^2}{M(\varphi_*)^{1/2}M_{Pl}^{3/2}T_{eq}} \sim \sqrt{\frac{M^2}{M(\varphi_*)\,\text{eV}}} \left(\frac{\varphi_*}{10^{11}\,\text{GeV}}\right)^2. \tag{4.70}$$

In particular, if the initial field value is roughly Planckian, $\varphi_* \sim 0.01 M_{Pl}$, the observed DM density is reproduced for ultra-light DM, $M \sim H_{eq} \sim 10^{-20}\,\text{eV}$, which would then lead to the observable smoothening of the galactic DM density profiles on scales comparable to the de Broglie wave-length, discussed in section 3.4.

In general, the DM density $\rho_\varphi$ inherits adiabatic inhomogeneities from the metric fluctuations. As discussed in the previous sections, the DM scalar $\varphi$, if lighter than $M \lesssim H_{infl}$, also undergoes non-adiabatic inflationary fluctuations of the order $\delta\varphi \sim H_{infl}$ in the inflationary deSitter space with Hubble constant $H_{infl}$. In this way its energy density $V(\varphi) \sim M^2\varphi^2/2$ acquires iso-curvature perturbations. The bound in eq. (1.37) demands that on cosmological scales the iso-curvature inhomogeneities are significantly smaller than the adiabatic inhomogeneities, so that $\delta\varphi \ll \varphi_*$ is needed. The iso-curvature bounds are bypassed if the state $\varphi$ only appears in the later stages of inflation: for example, it could be the pseudo-Goldstone boson of a symmetry that gets broken during inflation.

A scalar with a mass around the critical value $M = \frac{3}{2}H_{infl}$ needs a more complicated dedicated computation. The result is quite interesting [158]. Inflationary fluctuations get suppressed by a factor $e^{-4N_{infl}M^2/3H_{infl}^2}$ where $N$ is the number of $e$-foldings, equal to around 60 at the end of inflation. The iso-curvature perturbations are therefore suppressed on the cosmological scales, corresponding to $N \sim 60$. A scalar with $M \approx H_{infl}$ is thus a viable DM candidate. Based on eq.s (4.64) and (4.70), the cosmological DM density is reproduced for $H_{infl} \sim 2\,10^8\,\text{GeV}$.

A related possibility is worth mentioning. The inhomogeneities present during the Big Bang affect gravity leading to particle production, including DM production [159]. The resulting abundance is proportional to $\delta_k^2$, measured to be $\sim 10^{-9}$ at cosmological scales. An interesting DM abundance can arise if $\delta_k$ gets much bigger at smaller scales, a possibility discussed in section 4.6.1 in the different context of black hole production. Furthermore, DM could be produced from gravitational waves [160].

Let us mention in passing a different mechanism that may produce a scalar DM $\varphi$ with an average energy lower than the temperature of the plasma, so that the eV-scale DM could be cold enough. If the decay of a heavier particle or some other process starts producing a smooth spectrum of $\varphi$ quanta, the modes with lower momentum would generically have larger occupation numbers due to their larger wavelengths. Once the occupation numbers reach $\sim 1$, this results in a stimulated production of more of such low-energy quanta [161].

### 4.3.5   Dark Matter as super-heavy particles

The thermal relic abundance of stable particles heavier than about 100 TeV would be larger than the observed cosmological DM abundance, see eq. (4.26). Superheavy particles with a mass above this threshold, sometimes called **WIMPzillas**, can therefore be viable DM candidates only if they have at best very feeble interactions with the SM particles in the plasma. The small couplings and the large mass make it difficult to test these DM candidates experimentally. Such DM can be produced via many of the mechanisms discussed in previous subsections, where the main production channel depends on the details of the DM model. Among the many possibilities, let us highlight the following few:

○ In general the DM mass $M(\phi)$ depends on the inflaton vacuum expectation value $\phi$. This leads to an enhanced production of DM, if $M(\phi)$ crosses the zero when the inflaton oscillates during the period of reheating [155].

○ In certain unique theoretical models, DM can have an interaction of the form DM $X^3$, where $X$ is some complex particle. For DM mass in the range $m_X < M < 3m_X$ the DM particle is stable due to kinematic restrictions, while the DM $X^* \leftrightarrow XX$ scatterings together with the $XX^\dagger \to$ SM annihilations allow to reproduce thermally the observed cosmological DM abundance up to DM masses $M \sim 10^9$ GeV [162].

○ In order to minimize the model dependence, one can assume that DM is a particle with only gravitational interactions [154] (see section 9.8).

## 4.4   Asymmetric DM

An interesting question is whether the observed *baryonic* matter abundance could simply be a thermal relic. That is, can the thermal relic abundance estimate in eq. (4.9) also be applied to protons? Inserting the proton mass $M = m_p$ and the annihilation cross section $\sigma \sim 1/m_\pi^2$ gives proton and anti-proton relic abundances that are about 8 orders of magnitude smaller than the observed cosmological proton abundance. This means that the baryonic matter is *not* a thermal relic. As is well known, protons exist today for a different reason: a small primordial excess of protons over anti-protons, $\eta_B \equiv (n_B - n_{\bar{B}})/n_\gamma = (6.15 \pm 0.25)\, 10^{-10}$. This slight overabundance of protons was preserved until the present day because the baryon number is conserved. When the universe cooled below $T \sim m_p$, protons efficiently annihilated with anti-protons, and only the proton excess remained.

The origin of the primordial baryon asymmetry is not known, and various plausible mechanisms have been proposed. Following Sakharov [163], a baryon asymmetry is generated if, during some phase of the

cosmological evolution: 1) the baryon number is broken, 2) the CP symmetry is broken, and 3) an out-of-equilibrium dynamics takes place. A concrete example is baryogenesis via leptogenesis: right-handed neutrinos $N$ with masses $M \sim 10^{10}$ GeV decay via the same Yukawa couplings $y_\nu NLH$ that also generate the small SM neutrino masses.[16] Such decays are out-of-equilibrium, if the decay rate $\Gamma \sim y_\nu^2 M$ is slower than $H \sim T^2/M_{\rm Pl}$ at $T \sim M$. The Yukawa couplings contain CP-violating phases, which are essential for creating CP asymmetric decays of heavy sterile neutrinos, so that more leptons than anti-leptons are created. This produces a lepton asymmetry $\mathcal{L}$, which finally generates a baryon asymmetry $\mathcal{B}$. The last step relies on the fact that in the SM both the baryon and lepton numbers have weak anomalies, which give rise to the so called sphaleron processes that induce rapid violations of $\mathcal{B}$ and $\mathcal{L}$ at $T \gtrsim T_{\rm eq}$. Below the critical temperature $T_{\rm ew} \sim v$, at which the weak symmetry is restored, both $\mathcal{B}$ and $\mathcal{L}$ are (to an extremely good approximation) preserved. Taking into account that $\mathcal{B} - \mathcal{L}$ and electric charge $Q$ are conserved, the thermal equilibrium conditions imply that the two satisfy $\mathcal{B}/\mathcal{L} = -28/51$. Therefore, a generation of $\mathcal{L}$ implies a formation of $\mathcal{B}$ of similar size (see [164] for reviews of this and other mechanisms).

The above lengthy introduction is relevant for DM, since, if DM $\neq \overline{\rm DM}$, a similar mechanism may also generate the cosmological DM abundance. This would then be due to the asymmetry in DM and $\overline{\rm DM}$ abundances in the early Universe. For example, the right handed neutrino might have a CP-violating decay channel into some DM particles. While the symmetric relic contribution is annihilated away (and let us stress that, in order for that to happen efficiently, DM must have an annihilation cross section larger than in eq. (4.13)), the asymmetric contribution does not get annihilated away and constitutes the relic adundance [165]. In general terms, asymmetric DM point to two special values for the DM mass, $M \sim 5\,{\rm GeV}$ or $M \sim 2\,{\rm TeV}$:

◄ The first arises from assuming that the DM asymmetry $\eta_d$ is similar to the baryon asymmetry $\eta_B$. The predicted DM density $\Omega_{\rm DM} = \rho_{\rm DM}/\rho_{\rm cr} = M(n_{\rm DM} - n_{\overline{\rm DM}})/\rho_{\rm cr} = Mn_\gamma \eta_d/\rho_{\rm cr}$ would then be in the right (observed) ratio with the density of baryonic matter, $\Omega_{\rm DM} \approx 5\Omega_b$ (see eq. (1.1) and around), if the DM mass is $M \sim 5m_p$. No DM particle has been observed around this mass so far, which means that in viable models of this type the DM particle must have small interactions. A possible generation mechanism for both the baryon asymmetry and the DM abundance could be rare CP-violating decays of the SM mesons such as $B^0_{d,s}$, a possibility that has been realized in speculative models with complex enough dark sectors populated by the decays of the inflaton field (see Elor et al. (2018) in [165]). Another possibility for the decaying states are the sterile neutrinos, see section 9.7.

Relaxing the assumption $\eta_d \approx \eta_B$ allows of course to recover the observed DM abundance with arbitrary values of $M$, losing however the motivation of the connection with baryonic matter production.

▶ The second special value for the DM mass arises from assuming that the baryon and DM abundances are equilibrated by weak sphalerons, rather than by some new interaction. This happens, e.g., if DM is a bound state of fermions that are chiral under ${\rm SU}(2)_L$. If DM is mildly heavier than the critical temperature $T_{\rm ew}$, below which the weak sphalerons turn off, then the asymmetric DM abundance is suppressed by a Boltzmann factor, $\Omega_{\rm DM}/\Omega_m \approx e^{-T_{\rm eq}/M} M/m_p$. This suppression conspires with $M/m_p \gg 1$ to match the observed abundance, if $M \approx 8T_{\rm EW}$ (see, e.g., Nardi et al. in [166]).

Further discussion of the models underlying asymmetric DM can be found in section 9.7.

---

[16]Integrating out the heavy neutrinos gives a dimension 5 Majorana mass term for the SM neutrinos, $\mathcal{L}_{\rm eff} \sim (y_\nu LH)^2/m_N$. This gives SM neutrino masses that are suppressed by $v^2/M$ (the see-saw mechanism). Here $L$ is the SM lepton doublet, $H$ the Higgs doublet, and $v$ the electroweak vev of the Higgs. The generational indices are suppressed.

The phenomenology of asymmetric DM is discussed in the next chapters alongside the case of symmetric DM. A characteristic feature of asymmetric DM is that it does not annihilate. This means that there are no indirect detection signals associated with asymmetric DM. Asymmetric DM can accumulate in stars, affecting their evolution, and (more importantly) in white dwarfs, triggering their collapse into black holes. This implies a bound on the DM/nucleon interactions, at the level of direct detection bounds, see section 6.10.3.

Let us also mention in passing the existence of the literature on **partially asymmetric DM** scenarios [165], which interpolate between the symmetric and asymmetric cases. In these models, the population of $\overline{\mathrm{DM}}$ is not fully washed away (e.g., because the annihilation cross section is not large enough), or is restored via some external mechanism (e.g., via $\mathrm{DM} \to \overline{\mathrm{DM}}$ conversions). These scenarios typically point to a DM mass around the EW scale, and are thus also known as the *asymmetric WIMP* scenarios. Their phenomenology is a hybrid of the symmetric and asymmetric DM phenomenology. In particular, they feature DM annihilations, due to the residual symmetric population.

## 4.5   DM production from phase transitions

In several scenarios, phase transitions in the early Universe can play a role in producing the DM relic abundance, either in the form of dark monopoles (section 4.5.1) or in the form of particle DM (sections 4.5.2 and 4.5.3).

### 4.5.1   Dark magnetic monopoles

Some phase transitions, associated with the breaking of a symmetry group $\mathcal{G}$ to its subgroup $\mathcal{H}$, $\mathcal{G} \to \mathcal{H}$, can leave topologically stable field configurations known as *magnetic monopoles* [167] (see section 9.6, for more details). These are acceptable DM candidates, and their abundance can be estimated as follows.

The $\mathcal{G} \to \mathcal{H}$ phase transition that occurs at temperature $T$ results in about one relic monopole per correlation length volume, $\sim \xi^3$, i.e. the number density is $n \approx 1/\xi^3$. Since the correlation length $\xi$ must be smaller than the Hubble horizon $1/H$, this means that there is at least one monopole created per Hubble volume [167]. Parametrizing the correlation length as $\xi = \epsilon/H$, the number abundance $Y = n/s$ of DM monopoles normalized to the entropy density $s$ is then given by

$$Y \sim 1/(T\xi)^3 \sim (T/\epsilon M_{\mathrm{Pl}})^3. \tag{4.71}$$

A first-order phase transition results in $\epsilon \sim 1$, in which case the observed DM abundance, $Y \sim 0.4\,\mathrm{eV}/M$ (see eq. (4.3)), is reproduced for

$$M \sim (0.4\,\mathrm{eV} \times M_{\mathrm{Pl}}^3)^{1/4} \sim 10^{11}\,\mathrm{GeV}. \tag{4.72}$$

During a second-order phase transition the mass of a Higgs-like scalar crosses zero, giving correlation length $\xi \sim (M_V/H)^p/M_V$ where $p$ is the critical exponent, equal to $p = 1/3$ in the Ginzburg-Landau model [167]. This corresponds to $\epsilon \sim (H/M_V)^{1-p}$. Assuming gauge coupling $g \sim 1$, the observed DM abundance is reproduced for

$$M \sim M_{\mathrm{Pl}}(0.4\,\mathrm{eV}/M_{\mathrm{Pl}})^{1/(1+3p)} \sim 100\,\mathrm{TeV}, \qquad \text{for} \ \ p = 1/3. \tag{4.73}$$

## 4.5.2   Bubble collisions

In cosmological first-order phase transitions, the expanding walls of the nucleating bubbles can reach ultra-relativistic velocities. Wall interactions with the bath can then accelerate particles to high energies and accumulate them into shells. When the shells of two neighbouring bubbles crash into one another, the high-energy particles collide and can produce new particles that have a mass well above the scale of the phase transition and also above the highest temperature ever reached by the thermal plasma (a process called **the bubbletron**). If these new particles are the DM, then the bubble collisions are a version of a freeze-in mechanism for producing very heavy DM: typical results range from TeV-scale up to almost the Planck scale [168].

Another, different scenario for the production of DM involving the expansion of bubble walls is **filtered DM** [169]. In this framework, the DM particle acquires a mass during a cosmological first order phase transition. The expanding bubble walls act as a filter: only particles with sufficient momentum to overcome their mass inside the bubbles can pass through the walls; the others are reflected and quickly annihilate away. When the bubbles merge the transition is complete. The particles that have succeeded in penetrating the bubbles constitutes the relic DM today. Their abundance gets suppressed by a factor $e^{-M/2\gamma_f T_f}$, where $\gamma_f$ and $T_f$ are the Lorentz factor and temperature of the incoming fluid in the bubble wall rest frame. This can reproduce the observed DM abundance. The plausible range of DM masses is wide, depending on the parameters of the transition, and can extend up to the Planck scale.

A variation is **compressed DM** [169]. In this case, one considers the confining phase transition of a dark $SU(N)$ gauge group, in presence of dark heavy quarks. As the bubbles expand (at non-relativistic speed, in this case) and merge, their walls push the quarks into smaller and smaller regions. Inside these regions, because of the increased density, the quark interactions re-couple: they either annihilate or bind with one another. The bound states that are dark colour-neutral can then cross the wall of the squeezing region and escape into the true-vacuum phase, eventually diffusing and constituting the DM relic density today. A similar mechanism can be at play in the formation of primordial black holes (see section 4.6.2) and more generally macroscopic DM (see section 10.5.2, where it is discussed in much further details).

## 4.5.3   Super-cooling

After the Big-Bang the universe might undergo a phase of *thermal inflation* at a temperature above the DM mass [170, 171]. Thermal inflation is an extra phenomenon (unrelated to the inflation assumed in cosmology) that can happen if, for example, a thermal potential traps some scalar in a false minimum. This delays the onset of a phase transition such that for a while the energy density is dominated by the scalar vacuum energy, producing an inflationary-like period.

The period of thermal inflation could be the explanation for why DM has a small, sub-thermal, relic density: the inflationary super-cooling suppresses the densities of all particles, including that of DM. After the inflationary period is completed, the universe re-heats and populates the particle densities. However, if the reheating temperature is below the DM mass, its density can remain suppressed down to the desired level (including extra freeze-in contribution for the case where the DM mass is not well above the reheat temperature).

This happens, for example, if DM is massless in the initial phase and only acquires a mass from the phase transition that terminates the inflationary super-cooling cosmological period. More precisely, the super-cooling ends at some temperature below the critical temperature of the phase transition, when the tunnelling rate from the thermal vacuum to a new true vacuum becomes faster than the Hubble rate. An interesting possibility is that the phase transition that ends the super-cooling is the QCD phase transition, so that the supercooling ends at $T \sim \Lambda_{\mathrm{QCD}}$: in such a case the DM abundance is reproduced for a DM particle with mass around the TeV scale [171].

Figure 4.4:   *The blue curve shows the power spectrum $P_\zeta(k) \approx 2.1\,10^{-9}$ observed on cosmological scales, $k \lesssim 1/\mathrm{Mpc}$. The DM abundance is reproduced as Primordial Black Holes if $P_\zeta$ grows up to $\sim 10^{-2}$ around the green region, as exemplified by the dashed curve. The shaded red regions are excluded by the CMB μ-distortions and by a too large PBH abundance that would result in a DM abundance above the observed one [173].*

## 4.6    Formation of DM as primordial black holes

Primordial black holes (PBH) can be the DM, if they lie in the appropriate mass ranges not excluded by observations, see section 3.1.1 and in particular eq. (3.6). Here, we review the possible PBH formation mechanisms [172]. The starting point are primordial density inhomogeneities, whose cosmological evolution was discussed in section 1.3.1. On cosmological scales, comparable to the co-moving horizon today, $k \sim H_0$, the inhomogeneities were initially very small, $\delta_k \equiv \delta\rho/\rho \sim 10^{-5}$. When the clustering process increased the matter inhomogeneities to $\delta_k(t) \sim 1$, dark matter and ordinary matter collapsed gravitationally, and formed galaxies. These large scale inhomogeneities, however, cannot result in black hole candidates for DM. The collapse of collision-less DM does not form black holes, as discussed in section 2.1.2. The dissipative collapse of ordinary matter, on the other hand, formed stars and black holes heavier than a solar mass. These 'astrophysical' black holes cannot be DM, due to observational constraints, see section 3.1. That is, explaining DM as a population of black holes requires the existence of extra black holes [86]. They need to be *primordial*, because they must have formed before the CMB and BBN epochs, at temperatures $T \gtrsim \mathrm{MeV}$.

Primordial black holes would have formed out of ordinary collisional matter, without the need for extra DM particles, if the primordial matter density inhomogeneities at the appropriate scales were large, $\delta_k \gtrsim 1$. This is still possible for scales $1/k$ that are much smaller than those being probed by the current cosmological observations. Gravitational collapses of roughly spherical over-dense regions would then lead to the creation of black holes.

One of the primary quantities of interest is the mass distribution of primordial black holes. When a large primordial inhomogeneity on a scale $k$ re-enters the horizon, $k \sim aH$, it produces black holes with a typical mass comparable to the total mass contained within the horizon. This happens at a temperature $T \sim kM_{\mathrm{Pl}}/T_0$, which follows immediately from $k \sim aH$, using the Hubble rate $H \sim T^2/M_{\mathrm{Pl}}$, as appropriate for the radiation-dominated era, $T \gtrsim 10^3\,T_0$, and then inserting $T \sim T_0/a$. Since the

horizon radius was $R \sim 1/H$ and the cosmological density was $\rho \sim T^4$, the typical black hole mass is

$$M \sim \rho R^3 \sim \frac{M_{\text{Pl}}^2}{H} \sim \frac{M_{\text{Pl}}^3}{T^2} \sim \frac{M_{\text{Pl}} T_0^2}{k^2}. \tag{4.74}$$

The expression in terms of $H$ can, equivalently, also be derived from the Schwarzschild radius $R \sim GM$. Numerically, one obtains

$$M \sim 10^{15}\,\text{g} \left(\frac{t}{10^{-23}\,\text{s}}\right) \sim 2\,M_\odot \left(\frac{\text{GeV}}{T}\right)^2 \sim 250\,M_\odot \left(\frac{10^5\,\text{Mpc}^{-1}}{k}\right)^2. \tag{4.75}$$

The relation (4.74) is used in fig. 4.4 to link the PBH mass $M$ (upper horizontal axis) to the scale $k$ (lower axis). A few illustrative numerical examples are as follows:

⊖ If gravitational collapse occurred at a temperature near the QCD phase transition, $T \sim \Lambda_{\text{QCD}} \sim m_p$, the scale entering the horizon would be $k \sim 1/\text{pc}$. This would result in the formation of PBHs with masses comparable to the Chandrasekhar limit, $M \sim M_{\text{Pl}}^3/m_p^2 \sim M_\odot$, i.e., of the order of the solar mass.

⊖ A collapse occurring just before BBN, at a temperature $T \gtrsim \text{MeV}$, would result in PBHs whose masses are comparable to those of super-massive BHs found at the centers of present day galaxies, $M \sim 10^5\,M_\odot$.

⊖ At the other extreme, the formation of PBHs at the Planck temperature leads to black holes with Planck masses $M \sim M_{\text{Pl}} \sim 10^{-5}\,\text{g}$, which then quickly evaporate away.[17]

⊖ Finally, the PBHs in the asteroid mass range, where PBHs could be all of the DM (see section 3.1), would be formed at temperatures $T \sim \text{PeV}$.

The observed DM abundance is reproduced if the fraction of the total cosmological density that collapses into primordial black holes equals $\wp \sim T_0/T \sim 10^{-18}$ (similarly to eq. (4.3)). It can be even smaller, if the BHs later accrete significant amounts of matter.

The fraction $\wp$ of the total mass that collapses into black holes can be computed in terms of $\delta_k$ along the lines of section 2.2.5, obtaining a result similar to eq. (2.17). From it one can find that the typical inhomogeneity $\sigma_R \sim \delta_k \gg 10^{-5}$ needs to be enhanced up to $\sim 10^{-2}$, as illustrated in fig. 4.4 in terms of the power spectrum $P_\zeta \sim \sigma_R^2$. The required small $\wp \ll 1$ is obtained from exponentially rare regions with over-densities large enough that the PBHs are produced. Thereby the PBH abundance is exponentially sensitive to the parameters of the particular model, and models mostly tend to predict a narrow peak in the PBH mass distribution, as well as clusters of nearby primordial black holes [95]. Note also that eq. (2.17) was computed in the Gaussian approximation, while the small-scale fluctuations relevant for the PBH production could be non-Gaussian.

In the following we discuss specific mechanisms that could lead to the large inhomogeneities on small scales that are required for efficient PBH production.

## 4.6.1 Primordial black holes from enhanced inflationary perturbations

The small primordial inhomogeneities observed on cosmologically large scales are usually interpreted as imprints of quantum fluctuations during inflation, as discussed in section 1.3.1. As illustrated in fig. 4.4, the same inflationary mechanism could, in principle, produce large inhomogeneities on small scales, which collapse and generate PBHs.

---

[17]Small primordial black holes could be relevant for DM in a different, indirect way: their evaporation could produce DM particles [174].

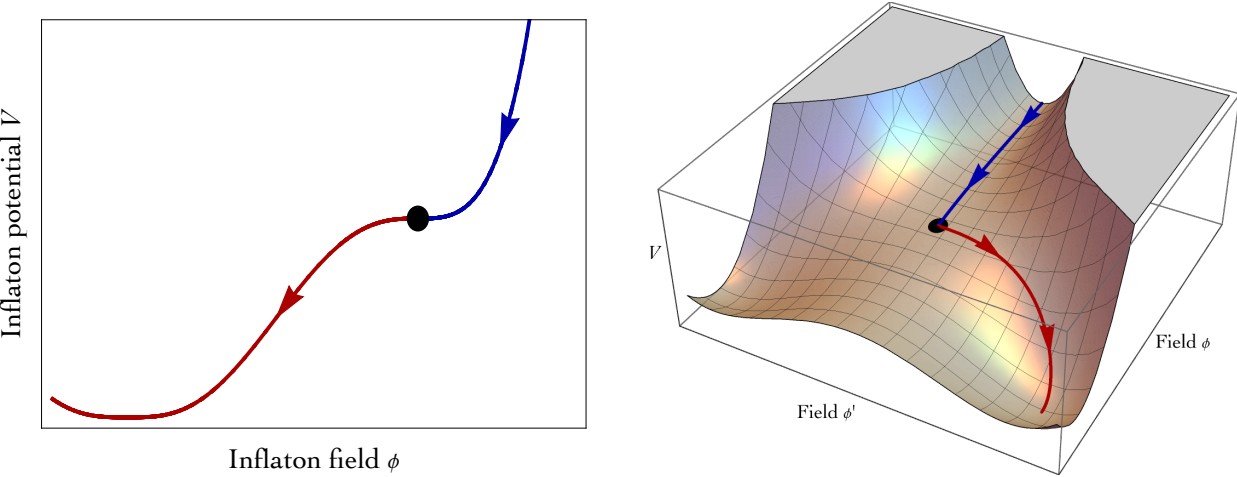

Figure 4.5: *Sample inflaton potentials that generate large inhomogeneities around the black dot along the inflationary trajectory.* **Left**: *one-field potential with an inflection point.* **Right**: *two-field water-fall hybrid inflation.*

As discussed in the previous section, PBHs form when the typical length scale of the large inhomogeneity matches the size of the Universe (i.e., the inhomogeneities on length scale $1/k$ re-enter the horizon). This occurs at some temperature $T$, and the corresponding Hubble rate $H$ then determines the resulting PBH mass via eq. (4.74). In the inflationary case under discussion, the horizon re-entering happens at $H \sim H_{\mathrm{infl}} \, e^{-2N}$, where $H_{\mathrm{infl}}$ is the Hubble constant during inflation and $N$ is the number of $e$-folds before the end of inflation. The corresponding temperature is $T \sim T_{\mathrm{RH}} e^{-N}$, where $T_{\mathrm{RH}} \sim \sqrt{M_{\mathrm{Pl}} H_{\mathrm{infl}}}$ is the reheat temperature (the temperature up to which the Universe reheats after the end of inflation). The above relations take into account that the scale factor expands exponentially during inflation, $a \propto e^{N}$, and the universe cools as $T \propto 1/a$ after inflation. Inserting this into eq. (4.74) shows that the typical PBH mass is

$$M \sim \frac{M_{\mathrm{Pl}}^2}{H_{\mathrm{infl}}} \, e^{2N}. \tag{4.76}$$

Assuming $H_{\mathrm{infl}} \sim 10^{14}$ GeV, eq. (4.76) then implies that the PBHs that have the mass in the observationally allowed range for them to be all of DM, $M \sim 10^{-12-16} M_{\odot}$, eq. (3.6), need to be seeded roughly $N \sim 20$ $e$-folds before the end of inflation (that is, roughly $\sim 30$ $e$-folds after the creation of cosmological inhomogeneities from quantum fluctuations).

This mechanism is testable. The power spectrum $P_{\zeta}(k)$ of primordial inhomogeneities unavoidably produces, at second order in cosmological perturbation theory, the so called Scalar Induced Gravitational Waves (SIGW) spectrum of gravitational waves, centered at present-day frequency $f = 2\pi k$ and with energy density [175]

$$\Omega_{\mathrm{GW}} \equiv \frac{1}{\rho_{\mathrm{tot}}} \frac{d\rho_{\mathrm{GW}}}{d \ln f} \sim 10^{-5} P_{\zeta}^2. \tag{4.77}$$

Inhomogeneities on small-scales large enough to seed primordial black holes in the DM allowed range, imply gravitational waves with Hz-scale frequency and energy density $\Omega_{\mathrm{GW}} \sim 10^{-9}$: a signal testable by future experiments such as LISA.

Which inflationary models lead to a nontrivial, scale dependent, primordial inhomogeneities, such as the example in fig. 4.4? Minimal inflation predicts a nearly scale-independent spectrum of primordial inhomogeneities (in agreement with data, though probed in a limited range of $k$ values, see solid and dotted blue lines in fig. 4.4). Non-minimal theories of inflation are therefore needed in order to have large

primordial inhomogeneities on small scales (at large values of $k$), and small inhomogeneities on large scales (at small values of $k$).

Large inhomogeneities arise when quantum fluctuations become almost comparable to the classical evolution of the inflaton field. This can happen, if quantum effects get large[18], or if classical effects get small. The latter is realised if the inflation potential has a feature that slows down the inflaton $\phi$, leading to an *ultra-slow roll* phase during which inhomogeneities $P_\zeta \approx V/24\pi^2 \bar{M}_{\rm Pl}^4 \epsilon$ are enhanced by the ultra-small parameter $\epsilon = (d\phi/dN)^2/2\bar{M}_{\rm Pl}^2$. The power spectrum can be computed precisely by solving for the evolution of the quantum modes $\delta\phi_k$ of the inflaton. The computation is simplified, if one uses the coordinate-independent combination $\zeta_k = -\Psi_k + \delta\phi_k/\dot\phi$, where $\Psi$ is the gravitational potential of eq. (1.20). The curvature perturbation $\zeta$ indicates the spatial curvature of surfaces of constant $\phi$. The $k-$Fourier mode of curvature perturbation, $\zeta_k$, determines the power spectrum at late times as $P_\zeta(k) = k^3 |\zeta_k|^2/2\pi^2$, and evolves following the Mukhanov-Sasaki equation [176]

$$\frac{d^2\zeta_k}{dN^2} + (3 + \epsilon - 2\eta)\frac{d\zeta_k}{dN} + \frac{k^2}{a^2 H^2}\zeta_k = 0, \qquad \eta = \epsilon - \frac{1}{2}\frac{d}{dN}\ln\epsilon. \tag{4.78}$$

Amplification happens for $3 + \epsilon - 2\eta < 0$, which can be satisfied during an ultra-slow roll phase with large $\eta \gtrsim 1$. Examples of features in the potential that lead to an amplification are an *especially flat* region such as an approximate plateau, an *inflection point* (either exact with $V' = 0$, or approximate with small $V'$), a *little bump* in the inflation potential that slows downs or possibly mildly traps the inflaton, or *enhanced dissipative friction* [177, 178].

In models with a single inflaton field, the required ultra-flat potential can be achieved via tuning of parameters or by assuming a pole in the inflaton kinetic term [177, 178]. Extreme flatness naturally arises in *'hybrid inflation'* models with two scalars $\phi$ and $\phi'$, if the potential has the *'water-fall'* form. Namely, we can assume that initially $\phi' = 0$ because its field-dependent squared mass $m_{\phi'}^2(\phi)$ is positive for initial values of $\phi$, and that the $m_{\phi'}^2(\phi)$ turns negative at some critical value $\phi_*$. As illustrated in fig. 4.5, at this point the potential is extremely flat along the $\phi'$ direction, and a phase transition happens. In general, phase transitions can be induced during inflation by the time-dependence of the inflaton field, and can thus also proceed in the reverse order (i.e., from a broken phase at early times to an unbroken phase at later times). For instance, the symmetry that keeps light some additional 'curvaton' field might get restored during later stages of inflation. The 'water-fall' model provides a specific realization of a quite general idea of *'double inflation'*, i.e., two separate periods of inflation, which generate different amounts of inhomogeneity, and on different scales.

A different possibility is a *sharp step-like feature* in the quasi-flat inflaton potential or a *dip*. In this case, the sudden acceleration makes the slow-roll parameter $\eta$ temporarily large, mildly enhancing fluctuations [177]. A larger effect can arise from a few consecutive steps or dips. Yet another possibility is that the inflaton fragments into localised solitons, termed *'oscillons'* [179]. An attempt at producing the PBHs within the SM Higgs inflation is discussed in section 9.1.6.

In most of these models an enhancement of $P_\zeta$ by 6 orders of magnitude needs a tuning of the model parameters with a precision of $10^{-3}$.

## 4.6.2 Primordial black holes from first-order phase transitions

An entirely different possibility is that the PBHs were created after inflation. This is possible, if the cosmological evolution of the early Universe included a *first order phase transition* at some temperature $T$, which ended after the bubbles of the lower energy vacuum nucleated at random places and times and then expanded until the true vacuum filled the entire space. The bubble nucleation rate per unit volume is given by $\gamma \approx T^4 (S_3/2\pi T)^{3/2} e^{-S_3/T}$ where $S_3(T)$ is the thermal bubble action. The nucleation

---

[18]This happens at super-Planckian field values, which, however, leads to eternal inflation.

happening at temperature $T$ therefore has $S_3/T \approx 4 \ln M_{\mathrm{Pl}}/T$. As an example, a thermal potential of a scalar $\phi$ with negligible cubic, $V(\phi) \approx (g^2 T^2/12)\phi^2/2 - \lambda \phi^4/4$, leads to $S_3/T \approx 5.4g/\lambda$. A first order phase transition can form PBHs via multiple mechanisms [180]:

◎ **Collisions of bubbles** generate PBHs in large enough quantities to be the DM, provided that

$$\alpha \equiv \frac{\Delta V - d(\Delta V)/d\ln T}{g_* \pi^2 T^4/30} \sim 1, \qquad \frac{\beta}{H} \equiv -\frac{d\ln\gamma}{d\ln T} \approx \frac{d(S_3/T)}{d\ln T} \sim 1. \qquad (4.79)$$

The first condition demands that the change in the energy density $\Delta V$ due to the phase transition is an $\mathcal{O}(1)$ fraction of the total cosmological energy density. The second condition demands that the inverse duration of the phase transition, $\beta \equiv d\ln\gamma/dt$, is comparable to the Hubble rate at the time at which the transition happens, $\gamma \sim H^4$. This condition is needed because a nucleation rate that grows too fast leads to many small bubbles. The same bubble collisions also generate gravitational waves, which can allow to test the scenario.

◎ **Late blooming regions in super-cooled** first-order phase transitions can form PBHs in the following way. Regions where the phase transition is completed get filled by radiation or matter, whose energy density red-shifts away. Regions where bubbles accidentally form at a later time, become relatively denser, because their energy density is dominated by the vacuum energy of the false vacuum, which does not red-shift away. Denser regions can become black holes. PBHs in the mass range where they can be the DM candidates arise, if the super-cooling starts at a temperature $T \sim 10^{4-7}\,\mathrm{GeV}$ [177].

◎ **Matter compression**. If the first order phase transition occurs in the presence of a particular type of particle dark matter, which cannot easily enter bubbles of the new phase, then the bubble expansion compresses the dark matter, forming macroscopic objects and possibly black holes [104]. Specific realizations of this mechanism are discussed in section 10.5.

In all of the above cases new physics is needed, since the SM does not predict any first-order phase transition.[19]

In addition to collisions of bubbles, the PBHs could be formed via other similar objects that may get created during a particular phase transition, for instance cosmic strings, domain walls [180] or oscillons [179]. Finally, PBHs might be a property of particle DM rather than an alternative to it: one can devise models of DM particles that cluster via some attractive long-range force and lead to the creation of PBHs.

---

[19]The QCD phase transition, even though it is a cross-over, can still play a role: the presence of many non-relativistic particles around the QCD scale implies a mildly reduced pressure (see fig. C.2 in the appendix), which enhances the growth of milder pre-existing inhomogeneities [177]. The critical density threshold $\delta_c$ for PBH formation gets reduced by about 10%. This mild reduction significantly enhances the PBH formation rate because, similarly to eq. (2.17), the collapsed fraction is exponentially sensitive to the density threshold for the collapse. This effect also means that the power spectrum of primordial inhomogeneities, which is enhanced and nearly scale-independent at small scales, would give a PBH spectrum peaked around the solar mass. This effect is therefore not relevant for PBH as DM, see Carr et al. (2021) in [177].

# Part II

# Dark Matter detection

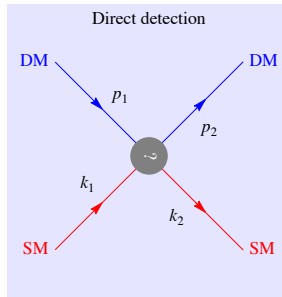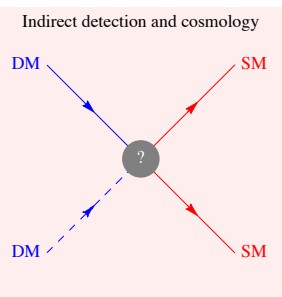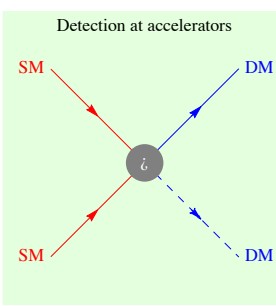

Figure 4.6: *The three main* **DM search strategies**, *from left to right: direct detection (the plot specifies the DM/nucleus kinematics used in chapter 5); indirect detection (chapter 6; one of the DM lines is dashed, indicating that both DM decays and DM annihilations are relevant); production at particle accelerators (chapter 7). The "SM" label stands for the Standard Model particle, or it can be viewed to stand for the "Standard Matter".*

Currently, the existence of Dark Matter is based on observations that only probe its gravitational coupling, see section 1. This gives us information about the total mass of DM in the Universe, as well as the amount of DM in a particular astrophysical region, including, to some extent, its spatial distribution. However, in order to really understand *what* Dark Matter is, we also need to observe its other interactions with ordinary matter.

The next question is: how?

There are three **main avenues of investigation** (traditionally schematized as in figure 4.6):

○ **Direct Detection** (Chapter 5) aims at detecting the recoil event produced by a passing DM particle hitting one of the nuclei or electrons in a highly shielded and closely monitored underground detector, made of ultrapure semiconductors, noble gasses, pristine crystals, etc.

○ **Indirect Detection** (Chapter 6) aims at detecting DM annihilations or DM decays, in our Galaxy or in other astrophysical environments, by searching for signatures in cosmic rays (in a broad sense: charged particles, photons or neutrinos arriving at Earth).

○ **Accelerator-based Detection** (Chapter 7) aim at producing DM particles in a controlled environment: e.g., in $pp$ collisions at the Large Hadron Collider at CERN, in $e^+e^-$ collisions at lower energy machines or in beam dump experiments[20], and then detecting the presence of DM via missing energy or other signatures.

In each of the above approaches there are many ongoing experimental efforts, with a good fraction of them represented in figure 4.7. They are distributed over many continents and are performed in very different environments. Most, if not all, of the experiments are run by international collaborations of varying sizes, from merely tens up to several thousands of scientists, from many institutions. The search for DM is a truly global effort of the international scientific community.

Chapters 5, 6 and 7 introduce the phenomenology of different search strategies as well as the computations relevant for predicting the expected signals, and discuss the current status. Chapter 8 discusses some of the recent observational anomalies.

---

[20]The same SM SM → DM DM processes also occur in astrophysical systems featuring energetic SM particles. However, the rates are expected to be too low to be phenomenologically relevant.

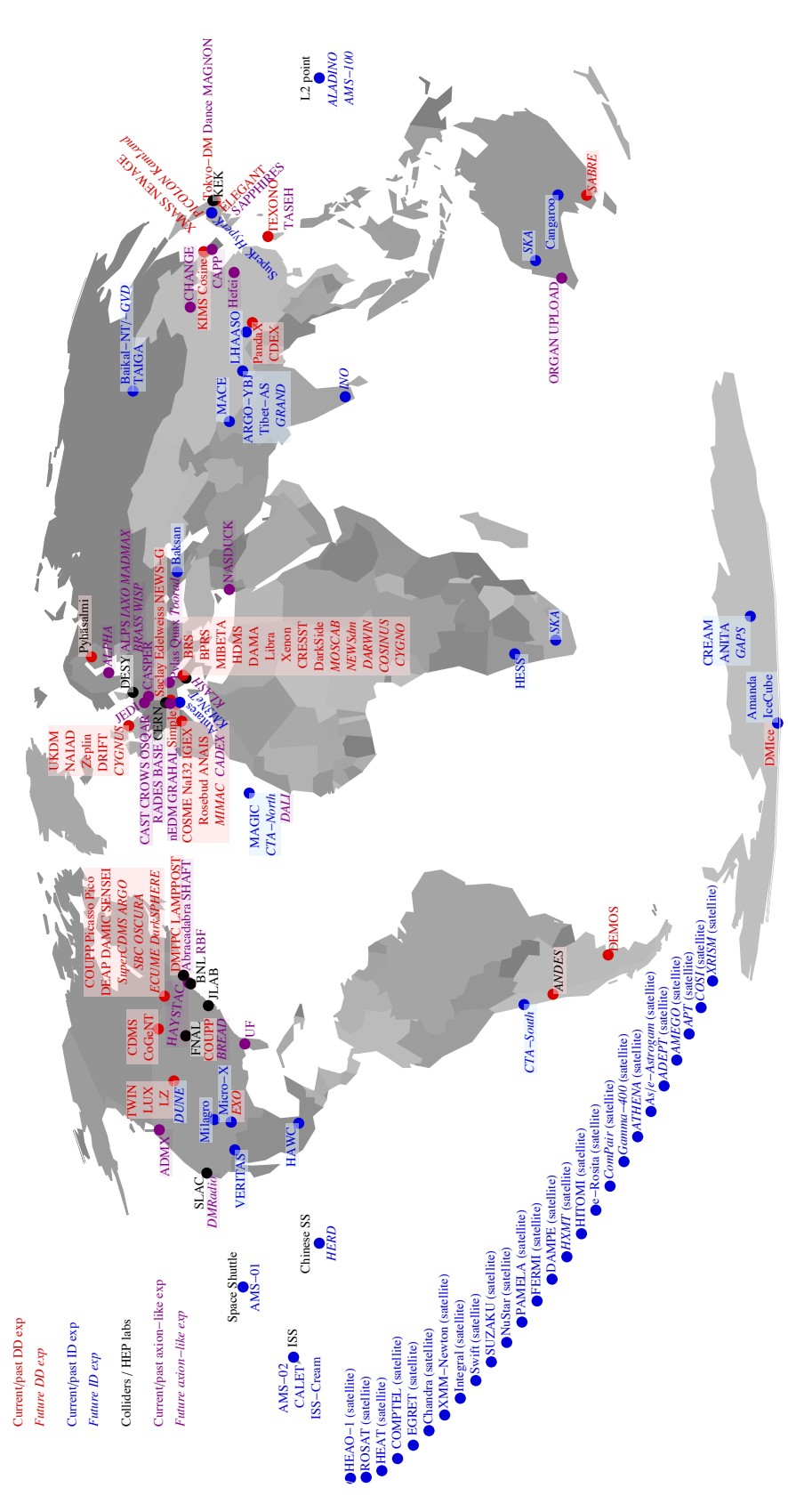

Figure 4.7: *The **experiments** trying to detect particle DM are performed all over the Earth (on surface as well as below ground and in orbit).*

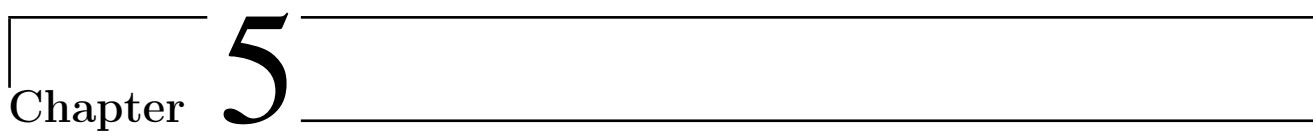

# Chapter 5

# Direct detection

Direct detection experiments aim to observe the energy that is deposited when a galactic Dark Matter particle scatters on the detector material. The Earth is well inside the DM halo so that there are DM particles constantly passing through the terrestrial detectors, and these could occasionally interact with the material in the detector. The main experimental challenges in detecting these events are that the DM scattering rates are very low (can be below one event per ton of target material per year) and that the deposited energies are small, typically anywhere from $\mathcal{O}(\text{keV})$ to $\mathcal{O}(\text{eV})$. In order to reduce the backgrounds the direct detection experiments are situated deep underground, and the detectors are made of ultra-pure materials.

In general, the signals of DM scattering in direct detection are of two types: due to DM scattering on either the atomic nuclei or on the electrons. The main features of the two types of scattering events are determined by kinematics. For simplicity, let us assume that the scattering is elastic and consider two simple examples, DM scattering on a free nucleus, $\chi A \to \chi A$, and DM scattering on a free electron $\chi e \to \chi e$. The energy $E_R$ transferred from DM to the target particle is comparable to the kinetic energy of the DM/target-particle two body system. This is given by $K = \mu v^2/2$, where $\mu = Mm/(M + m)$ is the reduced mass of the system with $m = m_A$ $(m_e)$ for scattering on the nucleus (electron). Furthermore, $v \approx 10^{-3}$ is the relative velocity, set by the typical velocity of DM in the galactic halo. So the typical momentum transferred to the target particle is $q \sim \sqrt{MK}$ in non-relativistic elastic collisions. The optimal situation is when the target particle has mass comparable to DM mass, $M \approx m$, see also fig. 5.4. This optimal situation gives:

| scattering on: | nucleus | electron |
|---|---|---|
| DM mass at which the experimental sensitivity is maximal: | tens of GeV | $\sim$ MeV |
| deposited energy, $E_R$: | few keV | few eV |
| momentum exchange, $q \equiv |\boldsymbol{q}|$: | tens of MeV | $\sim$ keV. |

There are of course many subtleties and exceptions to the above summary: DM could have a much faster component, scattering could be inelastic, DM could be absorbed instead of scattered, etc. We will cover these special cases in section 5.5. Also, for GeV or sub-GeV DM[1] the deposited energy is so small that one needs to consider the effects due to atoms and electrons being bound inside the material, which we discuss in sections 5.2, 5.3 and 5.4. Nevertheless, the main features are clear: in scatterings of DM on electrons the energy deposited in the detector will typically be smaller than for DM scattering on nuclei. This realization guides the requirements and designs of direct detection experiments, discussed in section 5.6.

---

[1]In direct detection literature quite often the sub-GeV DM is referred to as 'light DM'. We reserve the term light DM for sub-eV masses.

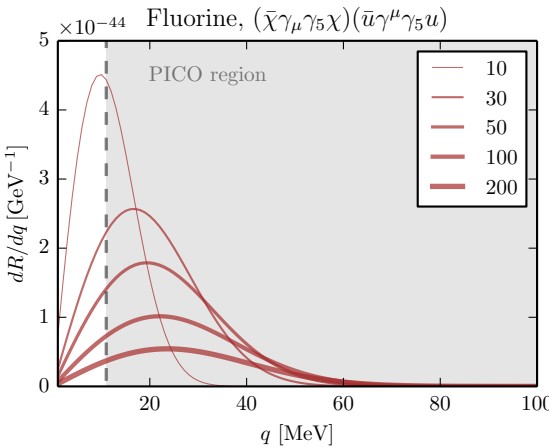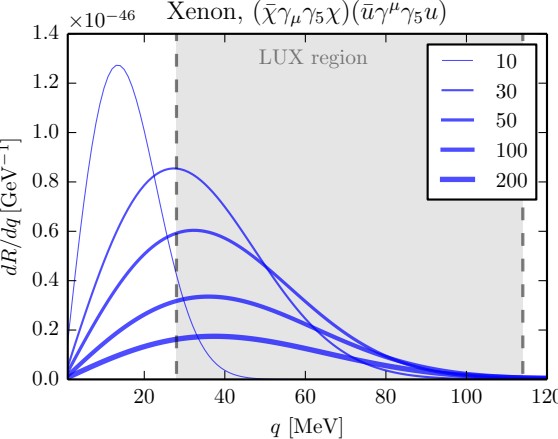

Figure 5.1:   *The **momentum exchange distributions**, dR/dq, for DM scattering rates on a representative light nucleus,* $^{19}$F *(left) and on heavy nucleus,* Xe *(right), for spin-dependent scattering. The approximate experimental thresholds are denoted by dashed vertical lines. Figure from [182].*

Historically, the first direct detection experiments in the late 1980s were an outgrowth of searches for neutrino-less double beta decays, since these experiments also searched for rare events, needed low backgrounds and were thus made of pure materials placed deep underground. In fact, even before these highly sensitive instruments were put in place, there were trivial constraints on strongly interacting DM from the mere fact that the NaI scintillators used for other purposes did not spontaneously "glow in the dark", as noted by Goodman and Witten in [181] (for more on history of DM direct detection see, e.g., Bertone and Hooper in [1]). For the next two decades the results of the experimental searches were almost exclusively interpreted in terms of DM scattering on nuclei. Below, we echo this historic progression and first focus on the DM–nucleus scattering, section 5.1, followed by the discussion of DM–electron scattering in section 5.2.

## 5.1   Scattering on nuclei

### 5.1.1   Main features and historic context

The DM direct detection experiments sensitive to the smallest event rates are at present based on DM/nucleus scatterings. Figure 4.6 (left) shows our kinematical conventions, with $p_{1(2)}$ and $k_{1(2)}$ the incoming (outgoing) four-momenta of the DM and the nucleus, respectively. The four momentum transfer to the nucleus is thus $q^\mu \equiv k_2^\mu - k_1^\mu$, while $q \equiv |\boldsymbol{q}| = |\boldsymbol{k}_2 - \boldsymbol{k}_1|$ is the magnitude of the three-momentum exchange. The kinematics of the collision are essentially the same as for the billiard balls when playing snooker: the incoming DM particle scatters on the nucleus that is initially at rest, and after the collision both the DM particle and the nucleus move. The largest momentum and energy transfer occurs when the DM particle and the nucleus have the same mass, and the collision is central. In that case the DM particle completely stops after the collision, giving all of its kinetic energy and momentum to the nucleus.

In general, the energy transferred from DM to the nucleus as measured in the center of mass frame, is an $\mathcal{O}(1)$ fraction of the kinetic energy of the DM–nucleus two body system, $K = \mu_A v^2/2$, where $\mu_A = M m_A/(M + m_A)$ is the reduced mass of the DM–nucleus system, and $v \approx 10^{-3}$ is the relative velocity. Numerically, $K \approx 20\,\mathrm{keV}$, if DM has a mass comparable to the heavy nuclei, $M \approx m_A \approx 100\,\mathrm{GeV}$. This is small enough that the scattering is elastic, i.e., that after collision the nucleus remains in the same state as before the collision. At the same time, $K$ is also large enough that the scattering event can be

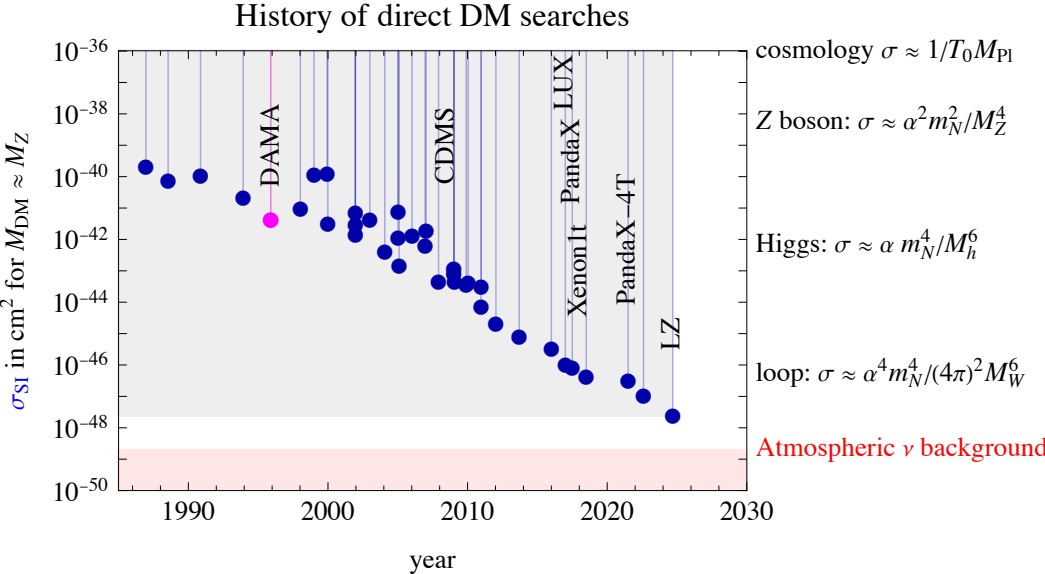

Figure 5.2: ***History of direct detection bounds*** *for spin-independent DM–nucleon scattering, setting the DM mass around the Z boson mass. Only the most stringent bounds (blue) at any given moment are shown, along with the claimed DAMA signal (section 8.1.1).*

detected.

More precisely, a typical momentum exchange during the collision is $q \sim 20$ MeV for scattering on light nuclei, such as F, and $q \sim 60$ MeV for scattering on heavy nuclei, such as Xe (see fig. 5.1). Experiments can tag the scattering events and reconstruct $K$ by observing at least one of the following three end-products: i) heat (phonons); ii) ionization; iii) scintillation.

The expected number of events per unit of time, assuming that DM particles have velocity $v$, is given by

$$\text{event rate} = N_T \frac{\rho_\odot}{M} v \sigma_A \approx \frac{1.5}{\text{yr}} \times \frac{m_T/A}{\text{kg}} \frac{\sigma_A}{10^{-39}\,\text{cm}^2} \times \frac{\rho_\odot}{0.4\,\text{GeV}/\text{cm}^3} \times \frac{v}{200\,\text{km/s}} \times \frac{100\,\text{GeV}}{M}, \quad (5.1)$$

where $m_T = N_T m_A$ is the mass of the detector composed of $N_T$ nuclei with atomic number $A$ and mass $m_A \approx A m_N$, where $m_N = 0.939$ GeV is the nucleon mass, and $\sigma_A$ is the DM cross section for scattering on the *nucleus*. For comparison, a typical cross section for particles that interact via QCD such as $p, \pi$ is $\sigma \approx 10^{-26}\,\text{cm}^2$, while for neutrinos of energy $E_\nu$ scattering on nucleons, a scattering that is mediated by the weak force, the cross section is $\sigma_{\nu N \to \nu X}/E_\nu \approx 10^{-38}\,\text{cm}^2/\text{GeV}$. We see that for weak scale cross sections one can expect only a few events per kg of material per year. Note that the above simple minded estimate has many caveats, which we will delve more into in the rest of this section. In particular, we assumed that the mean DM density in our local neighborhood is given by the value in eq. (2.11). However, inhomogeneities are expected, and thus the solar system could be in an over- or under-density (probabilities are estimated by Kamionkowski and Koushiappas in [183]), modifying the expected average scattering rates by orders of magnitude.

The DM cross section for scattering on a *nucleus*, $\sigma_A$, is usually converted to a cross section for DM scattering on a *nucleon*, $\sigma_N$. Quite often such scattering does not depend on nuclear spin. Then the main observable to be measured is the spin-independent cross section for DM scattering on nucleons, $\sigma_N = \sigma_{\text{SI}}$. This is related to the cross section for heavy DM scattering on nuclei as $\sigma_{\text{SI}} \approx \sigma_A/A^4$ (see eq. (5.11) below).

The historic progression of bounds on spin-independent scattering cross section is summarised in

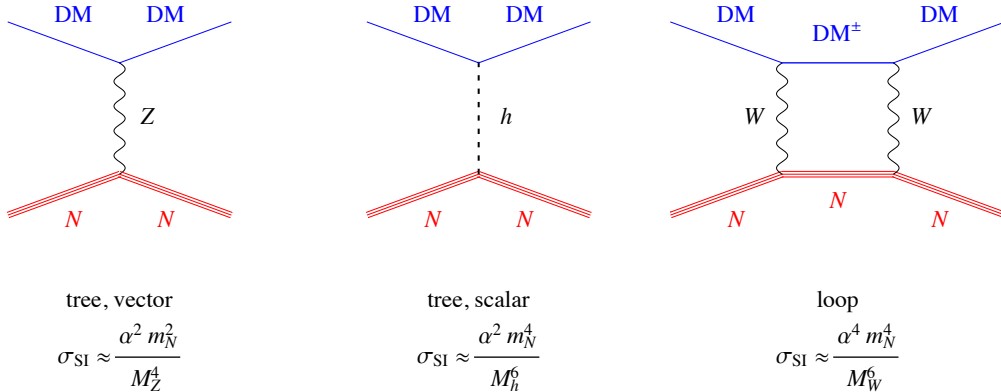

Figure 5.3: *Typical values of the **DM scattering** cross sections from tree level Z exchange (left), Higgs exchange (middle), and 1 loop electroweak correction (right).*

fig. 5.2, assuming for concreteness DM with mass $M \approx M_Z$, which roughly maximises the bounds. The most sensitive experiments at present, LZ [184], XENON1T [185] and PANDAX-4T [186], set a bound on the DM/nucleon cross section down to $\sigma_{\rm SI} \lesssim 10^{-46}\,{\rm cm}^2$, with the complete $M$ dependence of the exclusion shown in fig. 5.5. Figure 5.2 also shows the expected DM scattering cross sections for several examples of DM interactions:

- ▷ DM particles which undergo strong interactions would have $\sigma_{\rm SI} \sim 1/\Lambda_{\rm QCD}^2 \sim 10^{-26}\,{\rm cm}^2$. As discussed in section 5.1.2, such DM particles would have been stopped by collisions in the upper atmosphere and have been largely excluded.

- ▷ Tree-level $Z$ exchange would give $\sigma_{\rm SI} \sim \alpha_Y^2 Y_{\rm DM}^2 m_N^2/M_Z^4 \sim 10^{-38}\,{\rm cm}^2$ and is essentially excluded, unless DM has a small (effective) hypercharge $|Y_{\rm DM}| \lesssim 10^{-4}$.

- ▷ Tree-level Higgs exchange (with one loop coupling to gluons) would give $\sigma_{\rm SI} \sim \alpha_{\rm DM} m_N^4/M_h^6 \sim 10^{-43}\,{\rm cm}^2$ where $\alpha_{\rm DM} \sim y_{\rm DM}^2/4\pi$, if DM is a fermion with a Yukawa coupling $y_{\rm DM}$, presently constrained to be smaller than about 0.1.

- ▷ DM with $\mathrm{SU}(2)_L$ interactions and $Y = 0$ can scatter through 1-loop electroweak diagrams as in fig. 5.3, giving $\sigma_{\rm SI} \sim \alpha_2^4 m_N^4/M_W^6 \sim 10^{-45}\,{\rm cm}^2$. Computations find numerical factors of order $10^{-2}$ (see table 9.2), so this is allowed.

Experimental progress will become more difficult when experiments reach the background due to solar and atmospheric neutrinos, plotted as a grey region in fig. 5.5 (bottom) and discussed further in section 5.6.5.

## 5.1.2 Direct detection of strongly-interacting DM

In principle, DM particles could have large scattering cross sections with matter.[2] Such DM particles would hit the Earth with velocity $v \sim 10^{-3}$ (kinetic energy $E = Mv^2/2$) and get stopped by the Earth's atmosphere, without reaching the underground detectors. Heavy neutral particles loose energy in matter

---

[2]Usually, 'strong interactions' is synonymous with 'QCD interactions'. However, in the case of strongly interacting DM the large cross sections may be either due to QCD or due to new interactions. Note that 'strongly interacting massive particle (SIMP)' acronym has been used both for the case where DM has strong interactions with normal matter, as well as for the case where there are strong interactions within the dark sector but only feeble interactions with normal matter.

as [187]

$$\frac{dE}{dx} = -E \sum_A n_A \sigma_A \frac{2m_A}{M} = -2E\rho\langle A^4 \rangle \frac{\sigma_{\mathrm{SI}}}{M} \qquad \text{for } m_A \ll M, \tag{5.2}$$

where $n_A$ is the number density of nuclei with atomic number $A$ and mass $m_A \approx A m_N$, $2m_A/M$ is the fractional energy loss per collision, while $\sigma_A \approx \sigma_{\mathrm{SI}} A^2 (m_A/m_N)^2$ is the coherently enhanced cross section for DM scattering on a nucleus, with $\sigma_{\mathrm{SI}}$ denoting the spin independent scattering cross section for DM scattering on a single proton or neutron (assumed to be the same for simplicity, see also discussion in section 5.1.4 below). The number densities of targets in matter can be written as $n_A = f_A \rho / m_A$, where $\rho$ is the total mass density and $f_A$ the mass fraction of material $A$, such that $\sum_A f_A = 1$. Eq. (5.2) leads to the following expression for the energy loss of DM when passing through the material,

$$E(x) = E_0 \exp\left[ -\frac{\mathrm{TeV}}{M} \frac{\sigma_{\mathrm{SI}}}{\pi/\Lambda_{\mathrm{QCD}}^2} \frac{\int \rho \, dx}{7 \, \mathrm{kg/m^2}} \frac{\langle A^4 \rangle}{16.6^4} \right], \tag{5.3}$$

where for convenience we normalized the $\sigma_{\mathrm{SI}}$ to the typical QCD cross section, $\pi/\Lambda_{\mathrm{QCD}}^2 \approx 1.6 \cdot 10^{-26} \, \mathrm{cm^2}$. The Earth's atmosphere has a column depth $\int \rho \, dx = 10^4 \, \mathrm{kg/m^2}$ and $\langle A^4 \rangle^{1/4} \approx 16.6$. The crust has $\langle A^4 \rangle^{1/4} \approx 31$ and density $\rho \approx 3 \, \mathrm{g/cm^3}$. This explains why underground detectors cannot probe too large values of $\sigma_{\mathrm{SI}}/M$, as plotted in fig. 5.5 (top). Close to the upper boundary of the area excluded by the underground detectors (red shaded area in fig. 5.5 (top)), the strongly interacting DM particles undergo multiple interactions in detectors, a signature that is usually rejected in direct detection searches, and thus dedicated analyses of data are needed. This provides a *ceiling* to the reach of underground detectors, as pointed out by Albuquerque and Baudis (2003) and more recently refined by Kavanagh (2018) [187].

For larger cross sections, DM particles never reach the underground detectors. These larger values of $\sigma_{\mathrm{SI}}$ can be tested by a combination of DM searches performed using surface detectors, airborne experiments on balloons, rockets or satellites [187]. The extensive excluded areas are shaded in blue in fig. 5.5 (top). A related exclusion can be derived exploiting the fact that, upon multiple scatterings, heavy enough DM particles with galactic initial velocity $v \sim 10^{-3}$ slowly sink into the Earth, thermalizing with the atmosphere or the rock. Such a population of trapped, low-energy DM has been probed by collisions on excited tantalum: its nucleus has an excited state with angular momentum $\ell = 9$, which is so long lived that its decay has never been observed. DM would de-excite the nucleus, which then places a bound $\sigma_{\mathrm{SI}} \lesssim 10^{-31} \, \mathrm{cm^2}$ at $M \sim \mathrm{TeV}$ (Lehnert et al. (2020) in [187]). Another complementary exclusion (magenta region in fig. 5.5 (top)) arises from requiring that the measured terrestrial heat flow is not appreciably altered by the energy deposited in annihilations of DM particles that got captured by the Earth (see Mack et al. (2007) and Mack & Manohar (2013)). This bound therefore only applies to annihilating DM.

Very large cross sections (gray shaded area in fig. 5.5 (top)) are excluded by the absence of modifications in the CMB and the large scale structure power spectrum, since a large interaction between (cold) DM and baryonic matter would imply an excessive momentum transfer among the two species. This upper bound falls roughly at the same place as the constraints on DM self-interaction discussed in section 1.2.1. Slightly weaker constraints are obtained from the non-observation of a flux of $\gamma$-rays produced by the decay of neutral pions, which would have originated from collisions between the DM particles and the Galactic cosmic rays. BBN also imposes bounds, though orders of magnitude weaker than the ones above. All these bounds hold up to $M \sim 10^{16} \, \mathrm{GeV}$, above which the DM flux becomes too small to be observable by the current DM detectors. A dedicated analysis using the massive DEAP detector reaches almost the Planck mass.

Finally, the bounds discussed here apply to *spin-independent* (SI) DM/matter scattering (see section 5.1.4). A subset of analogous constraints for the *spin-dependent* (SD) scatterings (see section 5.1.5) have

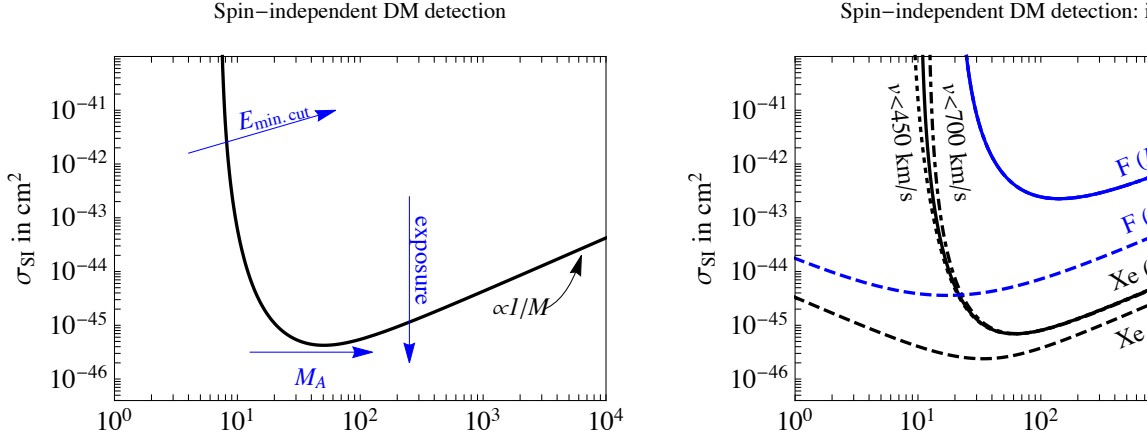

Figure 5.4: **Left**: *a **typical exclusion curve** from a direct detection experiment, with cross sections and DM masses above the curve excluded. **Right**: examples of exclusion curves for SI scattering on xenon (black curves) and fluorine (blue) for two choices of recoil energy cut, $E_R > 10$ keV (solid), and no energy cut (dashed), assuming no backgrounds. The variation on galactic escape velocity of the DM halo model, eq. (2.23), is only visible for low DM masses ($v_{\rm esc} = 450$ km/s, $550$ km/s, $700$ km/s are shown with dotted, solid and dot-dashed lines).*

also been computed, though a systematic analysis is still lacking[3]. As a rule of thumb, the bounds on SD interactions derived from nuclear scatterings are about 3 to 5 orders of magnitude looser than the corresponding SI bounds, since the latter benefit from the $A^2$ scaling with the nuclear mass number $A$ (see section 5.1.4). However, as pointed out by Digman et al. (2019), the constraints based on the scattering on *nuclei* at large values of the cross sections ($\sigma_{\rm SI} \gtrsim 10^{-32} - 10^{-27}$ cm$^2$) should be taken with care, as the $A^2$ scaling may fail. The constraints based on the scattering on *protons*, such as those from the CMB and the cosmic rays, are not affected by these considerations.

### 5.1.3 Direct detection of weakly-interacting DM: generalities

Next, we assume that DM particles interact weakly enough so that they reach the underground detectors. The event rate of DM scattering on nucleus, $DM + A \to DM + A$ (defined as the number of counts per time interval per target mass, $R_{\rm ev} = dN_e/dt \times 1/m_T$) is

$$\frac{dR_{\rm ev}}{dE_R} = \frac{N_T}{m_T} \int_{v>v_{\rm min}}^{\infty} \frac{d\sigma_A}{dE_R} v \, dn_{\rm DM}(\boldsymbol{v}), \qquad \text{with} \quad dn_{\rm DM} = \frac{\rho_\oplus}{M} f_\oplus(\boldsymbol{v}) d^3\boldsymbol{v}. \tag{5.4}$$

Here $N_T$ is the number of nuclei in the target of mass $m_T$, and $E_R \sim \mathcal{O}(\text{keV})$ is the recoil energy of nucleus. If there is more than one species of nuclei in the target, $N_T d\sigma_A/dE_R$ is replaced with a sum over different nuclei, $\sum_i N_{T,i} d\sigma_{A,i}/dE_R$. The integration is over the DM velocity in Earth's frame, $v$, with $\rho_\oplus$ the DM density at Earth's location. Normally this is the same as the DM density at the position of the solar system, $\rho_\oplus = \rho_\odot \approx 0.4$ GeV/cm$^3$, see eq. (2.11). The DM velocity distribution on Earth, $f_\oplus(\boldsymbol{v})$, is normalized such that $\int d^3\boldsymbol{v} f_\oplus(v) = 1$. It is obtained from the DM velocity distribution in the rest frame

---

[3]The bounds from ICECUBE on annihilating DM captured in the Sun (Albuquerque and Perez de los Heros (2010) [187]) and from the heating of Jupiter (Croon and Smirnov (2023) [188]) are relevant for the SD case. See also the discussion in section 6.10.

of the Galaxy through a Galilean transformation to the Earth's frame, see eq. (2.31). For further details on the DM velocity distribution see section 2.3.

In order to transfer a kinetic energy $E_R$ to the nucleus, the incoming DM needs to have a minimal velocity

$$v_{\min} = \sqrt{\frac{m_A E_R}{2\mu_A^2}}, \qquad \text{where} \qquad \mu_A = \frac{m_A M}{m_A + M} \quad \text{is the nucleus–DM reduced mass.} \qquad (5.5)$$

The integral over the DM velocity in eq. (5.4) is then from $v_{\min}$ up to the escape velocity from the Galaxy (transformed into the Earth's frame), which is about $\sim 550$ km/sec, see eq. (2.24). Typical DM velocity is $\sim 200$ to $300$ km/sec, see eq. (2.25).

In eq. (5.4) the particle physics is encoded in the DM–nucleus scattering cross section, $d\sigma_A/dE_R$. The description of this scattering is simplified by the fact that both the incoming DM and the nucleons inside the nucleus are non-relativistic, with typical velocities of $\mathcal{O}(10^{-3})$ and $\mathcal{O}(0.1)$, respectively. Ignoring velocity-suppressed terms there are only two possibilities for non-relativistic interactions: either the interactions involve nuclear spin, giving spin-dependent (SD) scattering, or they do not involve nuclear spin, resulting in spin-independent (SI) scattering. The UV models of dark matter more often than not lead to SI interactions. If the SI interactions are present, these usually give the dominant contributions to the DM-nucleus scattering rates because of the coherent enhancement. The predictions for the SI scattering are also under better theoretical control, so that we focus on these first, followed by the discussion of the SD scattering. The discussion of more general DM interactions, including velocity suppressed ones, is deferred to section 5.1.9.

## 5.1.4  Spin-independent scattering

The SI cross section for DM scattering on *nucleus* is given by (see, e.g., G. Jungman et al. in [1] or [189–191])

$$\frac{d\sigma_{\mathrm{SI}}^A}{dE_R} = \frac{m_A}{2v^2\mu_N^2} A^2 \sigma_{\mathrm{SI}} F^2(q), \qquad (5.6)$$

where $\sigma_{\mathrm{SI}}$ is the average cross section for DM scattering on *nucleons* and $F(q)$ the nuclear form factor; $A$ is the atomic mass number of the nucleus with charge $Z$; $\mu_N$ is the reduced mass of the nucleon/DM system, $\mu_N = m_N M/(m_N + M)$ (for simplicity we work in the isospin limit, $m_p = m_n \equiv m_N$); the momentum exchange is given by $q = \sqrt{2m_A E_R}$ where $m_A \approx A m_N$ is the mass of the nucleus. The SI scattering rate of eq. (5.4) becomes

$$\frac{dR_{\mathrm{ev}}}{dE_R} = \frac{A^2 \sigma_{\mathrm{SI}} F^2(q)}{2\mu_N^2} \frac{\rho_\oplus}{M} \eta(v_{\min}), \qquad \text{where} \quad \eta(v_{\min}) = \int_{v>v_{\min}}^{\infty} \frac{f_\oplus(\boldsymbol{v})}{v} d^3\boldsymbol{v}, \qquad (5.7)$$

with $v_{\min}$ given in eq. (5.5). Above, we assumed for simplicity that the target only contains a single species of nuclei. All the astrophysical uncertainties in predicting the DM scattering rates are contained in the local DM density $\rho_\oplus$ times a single function $\eta(v_{\min})$, encoding the dependence of the signal on the DM velocity distribution. These are then multiplied by the nuclear inputs such as the average cross section, $\sigma_{\mathrm{SI}}$, given by[4]

$$\sigma_{\mathrm{SI}} = \left(\frac{Zc_1^p + (A-Z)c_1^n}{Ac_1^{p(n)}}\right)^2 \sigma_{\mathrm{SI}}^{p(n)} \approx \frac{1}{4}\sigma_{\mathrm{SI}}^p \pm \frac{1}{2}\sqrt{\sigma_{\mathrm{SI}}^p \sigma_{\mathrm{SI}}^n} + \frac{1}{4}\sigma_{\mathrm{SI}}^n, \qquad (5.8)$$

---

[4]Note that since $\sigma_{\mathrm{SI}}^{p(n)} \propto (c_1^{p(n)})^2$, it does not matter whether $\sigma_{\mathrm{SI}}^p$ or $\sigma_{\mathrm{SI}}^n$ is used on the r.h.s. of the first equality in eq. (5.8).

where

$$\sigma_{\text{SI}}^{p(n)} = \frac{\mu_N^2}{\pi}(c_1^{p(n)})^2, \tag{5.9}$$

is the SI cross section for DM scattering on a single proton (neutron). Here, $c_1^p, c_1^n$ are the SI couplings of DM to non-relativistic protons and neutrons in the non-relativistic interaction Lagrangian (see also eq. (5.63) below),

$$\mathscr{L}_{\text{int}}^{\text{NR}} = \sum_{N=n,p} c_1^N 1_N 1_{\text{DM}}, \tag{5.10}$$

where $1_N$ ($1_{\text{DM}}$) counts the number of nucleons (DM particles). The second, approximate, equality in (5.8) assumes $A \approx 2Z$, which is a very good approximation for light nuclei, such as fluorine. The upper (lower) sign in eq. (5.8) is for the same (opposite) signs of $c_1^p$ and $c_1^n$ (for simplicity we assume that these are real). For $c_1^p = -c_1^n$ one can therefore have an almost complete negative interference for the scattering on light nuclei. In heavier nuclei such as xenon or tungsten there are up to 50% more neutrons than protons and thus the last relation in eq. (5.8) becomes less accurate. In short, $\sigma_{\text{SI}}$ in eq. (5.8) implicitly depends on the choice of target through $A$ and $Z$, which means that cancellations between $c_1^p$ and $c_1^n$ cannot be exact for all isotopes and elements.

The limit of small momenta exchanged between DM and nucleus, $q \to 0$, corresponds to the long wavelength limit or, equivalently, to the point particle approximation for the nucleus. In this limit DM couples coherently to all the nucleons inside the nucleus, and so the scattering is coherently enhanced by a factor $A^2$. This is readily seen from eq. (5.6) once the normalization of the nuclear form factor, $F(0) = 1$, is taken into account. In the limit of small momenta transfers, $q \to 0$, the integrated cross section for DM scattering on nucleus is

$$\sigma_{\text{SI}}^A\Big|_{q\to 0} = A^2 \sigma_{\text{SI}} \frac{\mu_{\chi A}^2}{\mu_{\chi N}^2} \approx \sigma_{\text{SI}} \begin{cases} A^2 & \text{for } M \ll m_N, \\ A^4/(1 + Am_N/M)^2 & \text{for } m_N \ll M < m_A, \\ A^4 & \text{for } M \gg m_A, \end{cases} \tag{5.11}$$

where the additional factor of $A^2$ in the scattering of heavy DM is due to the different available recoil energies,

$$E_{R,\text{max}} = 2\mu_{A(N)}^2 v^2 / m_{A(N)}, \tag{5.12}$$

for scattering on nucleus (nucleon). For heavy DM, $M \gg m_{N,A}$, we have $\mu_{\chi A}/\mu_{\chi N} \approx A$, while $\mu_{\chi A}/\mu_{\chi N} \approx 1$ for GeV or sub-GeV DM, $M \ll m_{N,A}$.

The decoherence for nonzero $q$ is reasonably well approximated by a Helm nuclear form factor $F(q) = 3e^{-q^2 s^2/2}[\sin(qr) - qr\cos(qr)]/(qr)^3$, with $s = 1$ fm, $r = \sqrt{R^2 - 5s^2}$, $R = 1.2A^{1/3}$ fm. The decoherence can be important for typical momenta exchanges when scattering on heavy nuclei. For instance, $F(100 \text{ MeV})\big|_{A=19} = 0.77$, $F(100 \text{ MeV})\big|_{A=132} = 0.35$, for scattering on fluorine and xenon, respectively. Note that eq. (5.6), where a single form factor describes the nuclear response to both scattering on neutrons and on protons, applies only in the isospin limit. The isospin-breaking corrections are typically small, at the percent level. If they are included, the single $F$ is replaced by three form factors, one each for response to scattering on protons and on neutrons, and one for the interference term.[5]

---

[5]If the difference between proton and neutron distributions in the nucleus is taken into account, eq. (5.6) becomes,

$$\frac{d\sigma_{\text{SI}}^A}{dE_R} = \frac{m_A}{2\pi v^2}\left[Z^2(c_1^p F^{(p,p)})^2 + (A-Z)^2(c_1^n F^{(n,n)})^2 + 2Z(A-Z)c_1^n c_1^p(F^{(n,p)})^2\right]. \tag{5.13}$$

In the isospin limit the three form factors are equal, and are the same as the one used in eq. (5.6), $F^{(p,p)} = F^{(n,n)} = F^{(n,p)} = F$. In [192] the definite isospin notation was used for the coupling coefficients, $c_i^{0(1)} = (c_i^p \pm c_i^n)/2$, along with squares of form factors of definite isospin, i.e., the so-called nuclear response functions, $W_M^{00}, W_M^{01}, W_M^{11}$, where

An experimental bound on the number of scattering events, $N_{\mathrm{ev}}$, implies an upper bound on $\sigma_{\mathrm{SI}}$, see eq.s (5.4), (5.6). A typical bound in $M, \sigma_{\mathrm{SI}}$ plane is shown in fig. 5.4 (left). The region above the curve is excluded. For large DM mass, $M \gg m_A$, the kinematics of the scattering is controlled by the mass of the nucleus, since this is the lighter of the two masses. This is the same as in the scattering of a bowling ball on a much lighter ping-pong ball — the momentum transfer is controlled by the relative velocity and by the mass of the ping pong ball only. In our expressions the DM mass enters through $v_{\mathrm{min}}$ dependence on $\mu_A$, and through the DM number density, $n_{\mathrm{DM}} \propto 1/M$. For heavy DM the reduced mass is approximately constant, $\mu_A \approx M_A$, so that for heavy DM the direct detection bound on $\sigma_{\mathrm{SI}}$ scales as $\propto 1/M$.

The exclusion in the low DM mass region is limited by the experimental lower cut on the nuclear recoil energy, $E_R > E_{R\mathrm{min}}$. Experiments impose such a cut either due to the threshold in the sensitivity of the detectors or due to rising backgrounds at low recoil energies. Lowering $E_{R\mathrm{min}}$ makes experiments sensitive to lower DM masses, which is easy to understand from the kinematics. The lighter the DM, the higher the minimal velocity for fixed $M_A$ and $E_R$. At some point, for lighter and lighter DM, $v_{\mathrm{min}}$ becomes larger than the escape velocity in the Galaxy ($v_{\mathrm{esc}} \approx 550$ km/s), transformed to the Earth's frame, and the event rate goes to zero. Typical DM mass for which this happens in the present experiments is in the range of a few GeV. This is illustrated in the right panel of fig. 5.4. Without a cut on $E_R$ the experiments could in principle be sensitive to arbitrarily small DM masses (dashed curves). The sensitivity to $\sigma_{\mathrm{SI}}$ is strongest for $M \approx M_A$, where the exact value depends on $E_{R\mathrm{min}}$ (compare the solid and dashed curves in the right panel of fig. 5.4).

Direct detection depends on relatively well-known DM astrophysical quantities: the DM density and the DM velocity distribution, $f_\oplus(\boldsymbol{v})$. In fig. 5.4 (right) we used the standard halo model with the escape velocity $v_{\mathrm{esc}} = 550$ km/s. Variation in $v_{\mathrm{esc}}$ affects the exclusion for low DM masses, signalling that this probes the tails of the DM velocity distribution, especially for the scattering on heavy nuclei. This issue is discussed further in section 5.1.8.

The current constraints on SI scattering cross section are shown in fig. 5.5. The upper panel considers a wider set of cross sections and DM masses, as discussed in section 5.1.2. For smaller $\sigma_{\mathrm{SI}}$, the lower panel in fig. 5.5 shows the current most stringent constraints from the underground detectors, using different materials. Note that direct detection experiments can exclude only DM cross sections that are small enough such that DM reaches the underground detectors (the upper boundary of the red excluded region in the upper panel in fig. 5.5). For clarity we do not plot this upper boundary of exclusion for individual experiments in the lower panel in fig. 5.5. For very large cross sections, above $\sigma_{\mathrm{SI}} \gtrsim 10^{33}\,\mathrm{cm}^2$ it is also hard, if not next to impossible, to write down explicit models, where the mediators, because of their large couplings to visible matter, are not already completely excluded by other searches (such as the monojet searches at the LHC, rare kaon decays, cooling of stars, etc [193]).

## 5.1.5   Spin-dependent scattering

If DM interactions with matter are predominantly through DM spin, $\boldsymbol{S}_{\mathrm{DM}}$, the description of the DM–nucleus scattering becomes more involved. In the non-relativistic limit there are two types of interactions

---

$\overline{W_M^{00,11}} = \kappa_A\big[Z^2 F^{(p,p)2} + (A-Z)^2 F^{(n,n)2} \pm 2Z(A-Z)F^{(n,p)2}\big]$ and $W_M^{01} = \kappa_A\big[Z^2 F^{(p,p)2} - (A-Z)^2 F^{(n,n)2}\big]$, with $\kappa_A = (2J_A+1)/4\pi$, and $J_A$ the spin of the nucleus. In the isospin limit, $W_M^{00} = \kappa_A A^2 F^2$, $W_M^{11} = \kappa_A(A-2Z)^2 F^2$, $W_M^{01} = -\kappa_A A(A-2Z)F^2$, which shows that in general $W_M^{00} \gg W_M^{01}, W_M^{11}$. In particular, for small $q$ we have, $W_M^{00} \simeq \kappa_A A^2$, $W_M^{11} \simeq \kappa_A(A-2Z)^2$, $W_M^{01} \simeq -\kappa_A A(A-2Z)$. Numerically, for fluorine $W_M^{00} \simeq 57$, $W_M^{11} \simeq 0.16$, $W_M^{01} \simeq -3$, while for instance for heavy elements such as the xenon isotope $^{131}_{54}$Xe the SI form factors are significantly larger, $W_M^{00} \simeq 5.5 \cdot 10^3$, $W_M^{11} \simeq 170$, $W_M^{01} \simeq -960$, as the result of coherent enhancement.

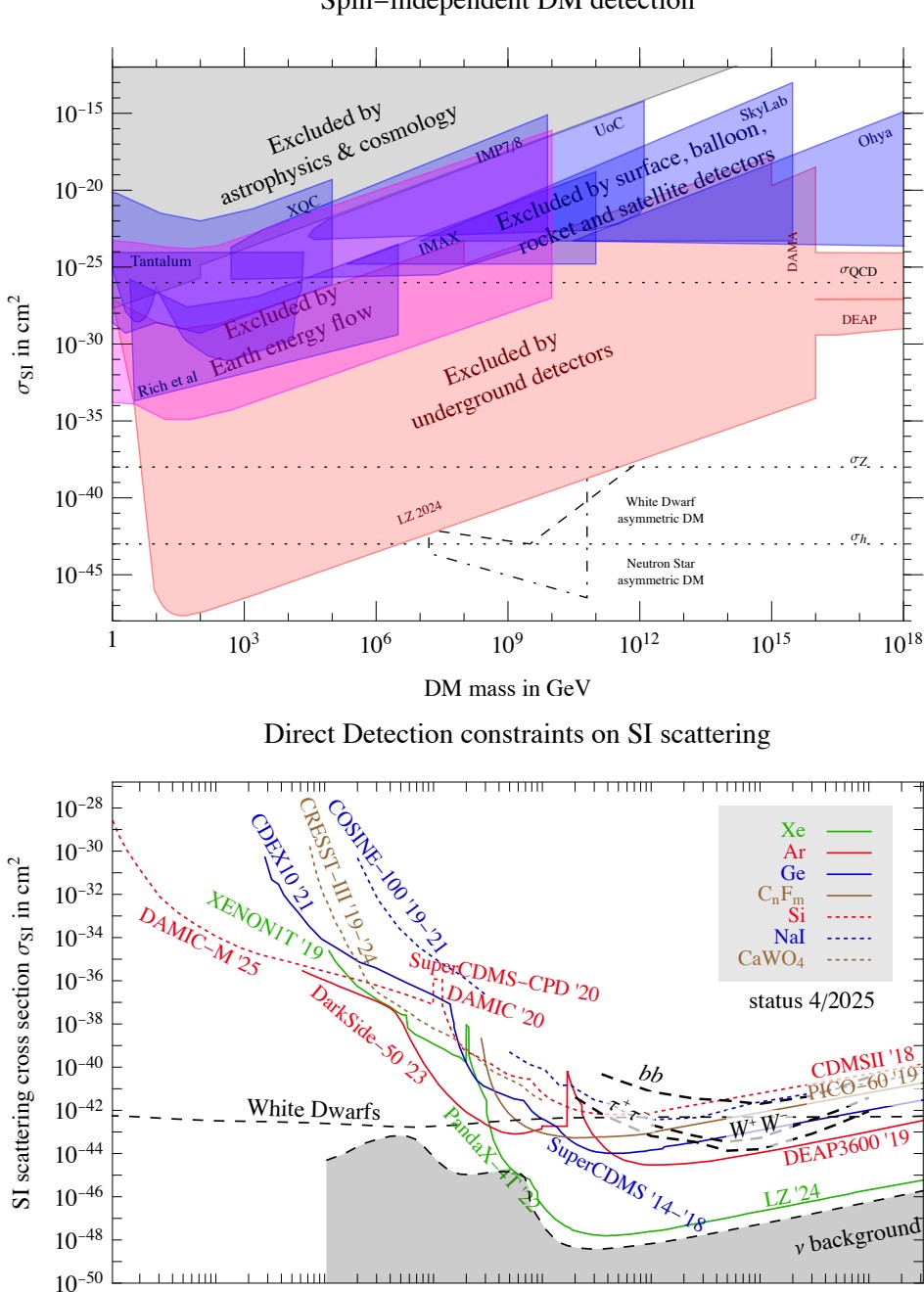

Figure 5.5: ***Bounds on the spin-independent cross section*** *from Dark Matter direct detection. The **top** panel shows a large range of cross sections, including the ranges for which DM does not reach the underground detectors, or leads to multiple scatterings (section 5.1.2 and [187]). The dashed (dot-dashed) black curves are bounds from White Dwarfs (Neutron Stars) applicable to asymmetric DM (see sections 6.10.3 and 6.10.4). The dotted horizontal lines indicate the typical QCD, tree level Z, or h-mediated cross sections. This figure effectively connects with fig. 3.4 at very large masses. The **bottom** panel shows the most stringent upper bounds, for different target materials, from the current underground detectors, table 5.2, rescaled using the local DM density in eq. (2.11). For clarity we do not indicate the upper ranges of exclusions, where for $\sigma_{SI} \sim 10^{-30}\ cm^2$ the sub-GeV DM does not reach even the surface detectors (upper panel). For large DM masses the quoted exclusions are naively linearly extrapolated. The dashed black lines show the bounds that require annihilating DM: from ICECUBE [194] (for three DM annihilation modes that produce neutrinos in the Sun, see section 6.9), and from White Dwarfs (see section 6.10.3). The neutrino background is discussed in section 5.6.5. The sub-GeV mass region is shown in further detail in fig. 5.16.*

between DM spin $S$ and the spin of nucleons, $S_N$ (we use the notation of [182, 192]),

$$\mathscr{L}_{\text{int}}^{\text{NR}} = \sum_{N=n,p} c_4^N \boldsymbol{S} \cdot \boldsymbol{S}_N + c_6^N \left( \boldsymbol{S} \cdot \frac{\boldsymbol{q}}{m_N} \right) \left( \boldsymbol{S}_N \cdot \frac{\boldsymbol{q}}{m_N} \right), \tag{5.14}$$

where $\boldsymbol{q}$ is the three-momentum transferred in the scattering. Its size is $|\boldsymbol{q}| \sim \mu_A v \sim 100$ MeV for scattering on heavy nuclei, and $|\boldsymbol{q}| \sim \mu_N v \sim 1$ MeV for scattering on a free nucleon (in each case assuming $M \gtrsim m_{A,N}$). The two non-relativistic coefficients, $c_4^N, c_6^N$, are affected differently by the QCD dynamics. For the case where heavy mediators couple DM to quarks, the $c_4^{p,n}$ coefficients do not depend on $q^2$ (the same as $c_1^{p,n}$ did not for the SI scattering). The $c_6^N(q^2)$, on the other hand, receive contributions from tree level exchanges of light mesons, $\pi^0$ and $\eta$, and are of parametric size $c_6^N \sim m_N^2/(m_\pi^2 + \boldsymbol{q}^2)$, see eq. (5.21) below.

For DM scattering on free nucleons or on light nuclei the contributions from $c_6^N$ are suppressed by $q^2/m_\pi^2$ and it suffices to include only the $c_4^N$ terms in eq. (5.14). This gives for the SD scattering of DM with incoming velocity $v$ on the unpolarized free nucleon:

$$\frac{d\sigma_{\text{SD}}^N}{dE_R} = \frac{\sigma_{\text{SD}}^N}{E_{R,\text{max}}}, \qquad \text{where} \qquad \sigma_{\text{SD}}^N = J_{\text{DM}}(J_{\text{DM}} + 1) \frac{(c_4^N)^2 \mu_N^2}{4\pi} \tag{5.15}$$

is the total cross section, and $E_{R,\text{max}}$ is the maximal recoil energy given in eq. (5.12), while $J_{\text{DM}}$ is the DM spin.

For SD scattering on heavy nuclei both terms in eq. (5.14) are of similar parametric size, giving the scattering cross section [192][6]

$$\frac{d\sigma_{\text{SD}}^A}{dE_R} = \frac{m_A}{6v^2} \frac{J_{\text{DM}}(J_{\text{DM}} + 1)}{2J_A + 1} \sum_{\tau\tau'} \left[ c_4^\tau c_4^{\tau'} W_{\Sigma'}^{\tau\tau'}(q) + \left( \frac{q^2}{m_N^2} \left( c_4^\tau c_6^{\tau'} + c_6^\tau c_4^{\tau'} \right) + \frac{q^4}{m_N^4} c_6^\tau c_6^{\tau'} \right) W_{\Sigma''}^{\tau\tau'}(q) \right]. \tag{5.16}$$

The sum runs over isospin, $\tau, \tau' = 0, 1$, where the non-relativistic coefficients are $c_i^{0(1)} = (c_i^p \pm c_i^n)/2$. The nuclear response functions $W_{\Sigma'}^{\tau\tau'}(q^2), W_{\Sigma''}^{\tau\tau'}(q^2)$ give the transverse and the longitudinal responses to the DM spin interactions. They take into account both the magnitude of the spin of the nucleus as well as the spatial distribution of the proton and neutron spins in the nucleus. In the nuclear shell model the SD nuclear form factors are expected to be larger, if the nucleus contains unpaired nucleons. Therefore the values of the SD form factors differ significantly between different isotopes of the same element. In the long wavelength limit, $q \to 0$, the two sets of nuclear response functions are proportional to each other, $W_{\Sigma'}^{\tau\tau'}(0) = 2W_{\Sigma''}^{\tau\tau'}(0)$, but differ at finite $q^2$. The values of the SD form factors are much more sensitive to the nuclear structure than the SI form factors are, and can differ substantially between different nuclear calculations (currently, the ab initio calculations are not possible for the heavy nuclei and thus various approximations are required). Note however, that the suppression of SD form factors with increasing $q$ is slower than for the SI form factors. Typically, the SD nuclear response functions are suppressed by a

---

[6]The other commonly used notation for the transverse spin form factors/nuclear response functions, see e.g. [191, 195], is $S_{00(11)}(q) = W_{\Sigma'}^{00(11)}(q)/4$, $S_{01}(q) = W_{\Sigma'}^{01}(q)/2$. In the isospin limit the $S_{\tau\tau'}$ and $W_{\Sigma'}^{\tau\tau'}$ can be expressed in terms of the proton spin averaged over the nucleus, $S_p$, and the averaged neutron spin, $S_n$. In terms of these, the $S_{\tau\tau'}$ nuclear form factors are given by $S_{00} = C(J_A)(S_p + S_n)^2$, $S_{11} = C(J_A)(S_p - S_n)^2$, $S_{01} = 2C(J_A)(S_p^2 - S_n^2)$, where $C(J_A) = (2J_A+1)(J_A+1)/(4\pi J_A)$, with $J_A$ the spin of the nucleus. This then gives $W_{\Sigma'}^{00} = 4C(J_A)(S_p+S_n)^2$, $W_{\Sigma'}^{11} = 4C(J_A)(S_p - S_n)^2$, $W_{\Sigma'}^{01} = 4C(J_A)(S_p^2 - S_n^2)$. For nuclei with an unpaired proton, $S_p \simeq 0.5$, $S_n \simeq 0$ (and vice versa for nuclei with an unpaired neutron). An example of a nucleus with an unpaired proton is the stable isotope of fluorine $^{19}_9$F that has spin $J_A = 1/2$, so that $W_{\Sigma'}^{00} \simeq W_{\Sigma'}^{11} \simeq W_{\Sigma'}^{01} \simeq 0.5$ in the long wavelength limit (a nuclear shell model calculation in [192] obtains $\approx 0.4$ for the three form factors). For the two stable xenon isotopes with nonzero nuclear spin, $^{129}_{54}$Xe with $J_A = 1/2$ and $^{131}_{54}$Xe with $J_A = 3/2$, one has both $S_p$ and $S_n$ nonzero and $W_{\Sigma'}^{\tau\tau'} \sim \mathcal{O}(0.1)$.

factor of a few at $q = 100$ MeV and by about two orders of magnitude at $q = 200$ MeV compared to their long wavelength values at $q = 0$.

If the $c_6^\tau$ contributions can be neglected, the SD scattering of DM on nucleus can be written in a form similar to the SI scattering, eq. (5.8),

$$\frac{d\sigma_{\rm SD}^A}{dE_R} = \frac{m_A}{2v^2\mu_N^2}\sigma_{\rm SD}S(q), \tag{5.17}$$

where $\sigma_{\rm SD} = \sigma_{\rm SD}^p + \sigma_{\rm SD}^n$, is the sum of cross sections for SD scattering on proton and neutron, eq. (5.15). The SD nuclear response functions are collected in

$$S(q) = \frac{1}{3(2J_A + 1)}\left[W_{\Sigma'}^{00} + W_{\Sigma'}^{11} + 2c_{2\theta}W_{\Sigma'}^{01} + s_{2\theta}\left(W_{\Sigma'}^{00} - W_{\Sigma'}^{11}\right)\right], \tag{5.18}$$

where $c_{2\theta} \equiv \cos(2\theta)$ and $s_{2\theta} \equiv \sin(2\theta)$ parametrise the dependence on the ratio of SD couplings of DM to neutrons and protons, $\tan\theta \equiv c_4^n/c_4^p$. The simplified form of the cross section in eq. (5.17) makes it easier to see that, unlike the SI cross section (5.11), the SD scattering is not coherently enhanced. In the $q \to 0$ limit the total SD scattering cross section is

$$\sigma_{\rm SD}^A\Big|_{q\to 0} = \sigma_{\rm SD}S(0)\frac{\mu_{\chi A}^2}{\mu_{\chi N}^2} \approx \sigma_{\rm SD}S(0) \times \begin{cases} 1 & \text{for } M \ll m_{N,A}, \\ A^2 & \text{for } M \gg m_{N,A}. \end{cases} \tag{5.19}$$

For GeV or sub-GeV DM the SD scattering cross section is given by SD scattering on individual nucleons, modified by the averaging over nuclear wave functions, encoded in $S(0)$. For heavy DM there is an extra $A^2$ factor due to the larger available kinetic energy. Note that compared to the SI scattering, eq. (5.11), the SD cross section does not have the additional enhancement due to the $A^2$ coherent factor. This gives the weaker experimental bounds plotted in fig. 5.6.

Next, let us quantify how good is the approximation of neglecting the $\sigma_6^\tau$ terms in eq. (5.16). We illustrate this using two examples of DM interactions, the axial vector currents and the tensor interactions. For simplicity we assume that DM is a Dirac fermion, and that only couplings to $u$ and $d$ quarks are nonzero (a more complete treatment, including other possible interactions, is deferred to section 5.1.9).

## DM with axial-vector interaction

Here, we assume that the effective DM interaction, which follows from integrating out some heavy mediators, takes the form (we use the notation and numerical values in [182])[7]

$$\mathcal{L}_{\rm int} = \sum_{q=u,d}\hat{\mathcal{C}}_{4,q}^{(6)}(\bar\chi\gamma^\mu\gamma_5\chi)(\bar q\gamma_\mu\gamma_5 q), \tag{5.20}$$

where the coefficients $\hat{\mathcal{C}}_{4,q}^{(6)}$ have dimensions of mass$^{-2}$ and encode the particle physics — the mass and the couplings of the heavy mediators (see also Section 5.1.6 below). The two non-relativistic coefficients in eq. (5.14) are

$$c_4^N = -4\sum_{q=u,d}\Delta q_N\hat{\mathcal{C}}_{4,q}^{(6)}, \qquad c_6^N = \frac{m_N^2}{m_\pi^2 + q^2}\sum_{q=u,d}a_{P',\pi}^{q/N}\hat{\mathcal{C}}_{4,q}^{(6)}, \tag{5.21}$$

---

[7]For Majorana fermion DM, $\chi_{\rm M}$, the definition of the Wilson coefficient includes and extra factor of $1/2$ , $\mathcal{L}_{\rm int} = \sum_{q=u,d}\frac{1}{2}\hat{\mathcal{C}}_{4,q}^{(6)}(\bar\chi_{\rm M}\gamma^\mu\gamma_5\chi_{\rm M})(\bar q\gamma_\mu\gamma_5 q)$. The factor of $1/2$ compensates the additional Wick contractions for the Majorana fermion, so that the expressions for the non-relativistic coefficients in (5.21) in terms of $\hat{\mathcal{C}}_{4,q}^{(6)}$ remain unchanged.

where $\Delta u_p = \Delta d_n \equiv \Delta u = 0.897(27)$ and $\Delta d_p = \Delta u_n \equiv \Delta d = -0.376(27)$ are the quark axial charges of proton and neutron (in the isospin limit), with $a_{P',\pi}^{u/p} = -a_{P',\pi}^{d/p} = a_{P',\pi}^{d/n} = -a_{P',\pi}^{u/n} = 2(\Delta u - \Delta d)$. For weak scale DM, $M \sim \mathcal{O}(100\,\mathrm{GeV})$, the momentum exchange is comparable to the pion mass, $q \approx m_\pi$, see fig. 5.1. The two contributions in eq. (5.14) are thus parametrically similar in size, with the $c_6^N$ term only suppressed by $q^2/m_\pi^2$ (note that the $m_N^2$ factor in (5.21) cancels the $1/m_N^2$ supression factor in eq. (5.14)).

We expect the $c_6^N$ terms to be numerically important for a heavy DM scattering on a heavy nucleus, such as xenon or tungsten. Ignoring $c_6^N$ term can then lead to an $\mathcal{O}(1)$ error in the predictions.[8] In contrast, for the scattering of GeV/sub-GeV DM or for the scattering on light nuclei, such as fluorine or oxygen, the $c_6^N$ terms can be safely ignored and one can use the simplified expression for the SD cross section, eq. (5.17).

**DM with tensor interactions**

If the DM interactions have the form

$$\mathscr{L}_{\mathrm{int}} = \sum_{q=u,d} \hat{\mathcal{C}}_{9,q}^{(7)} m_q \big(\bar{\chi}\sigma^{\mu\nu}\chi\big)(\bar{q}\sigma_{\mu\nu}\gamma_5 q), \tag{5.22}$$

the non-relativistic coefficients are

$$c_4^N = 8 \sum_{q=u,d} m_q A_{T,10}^{q/N}(0)\hat{\mathcal{C}}_{9,q}^{(7)}, \qquad c_6^N = 0, \tag{5.23}$$

with the generalized tensor form factors at zero recoil given by the quark tensor charges, $A_{T,10}^{u/p(d/n)}(0) = g_T^u = 0.794 \pm 0.0015$, $A_{T,10}^{d/p(u/n)}(0) = g_T^d = -0.204 \pm 0.08$. The tensor interactions thus lead to pure $\boldsymbol{S}_{\mathrm{DM}} \cdot \boldsymbol{S}_N$ interactions. They are an example of DM couplings for which the simplified expression for the SD cross section, eq. (5.17), applies to both the scattering of GeV/sub-GeV and heavy DM, and for scattering on either light or heavy nuclei.

## 5.1.6   Benchmark models for DM direct detection

We list benchmark models that lead to either SI or SD scattering, and the corresponding values for the DM/nucleon scattering cross sections, $\sigma_{\mathrm{SI}}^N$, $\sigma_{\mathrm{SD}}^N$.

*Z*-**mediated DM interactions**

The $Z$ boson might couple to DM so that the relevant part of the interaction Lagrangian is $\mathscr{L}_{\mathrm{int}} \supset -Z^\mu J_\mu$, where $J_\mu = J_\mu^{\mathrm{SM}} + J_\mu^{\mathrm{DM}}$ is the sum of the SM and the DM current,

$$J_\mu = J_\mu^{\mathrm{SM}} + \frac{g_2}{\cos\theta_{\mathrm{W}}} \begin{cases} \bar{\chi}\gamma_\mu(g_{\mathrm{DM}}^V + \gamma_5 g_{\mathrm{DM}}^A)\chi, & \text{if DM is a fermion } \chi, \\ g_{\mathrm{DM}}^S[S^*(i\partial_\mu S) - (i\partial_\mu S^*)S], & \text{if DM is a scalar } S, \end{cases} \tag{5.24}$$

with $g_2$ the weak coupling constant, and $\theta_W$ the weak mixing angle. The couplings $g_{\mathrm{DM}}^{V/A}$, $g_{\mathrm{DM}}^S$ depend on the details of the DM model. They are $\mathcal{O}(1)$ for DM that is part of an EW multiplet, and can be much

---

[8]For instance, the predictions for the scattering rates for scattering of DM with mass $M = 100\,\mathrm{GeV}$ on $^{131}\mathrm{Xe}$ are $\mathcal{O}(\{1\%, 30\%, 60\%, 20\%\})$ higher, if one ignores the $c_6^N$ terms, taking as the DM coupling benchmarks $\{\hat{\mathcal{C}}_{4,u}^{(6)} = \hat{\mathcal{C}}_{4,d}^{(6)}, \hat{\mathcal{C}}_{4,u}^{(6)} = -\hat{\mathcal{C}}_{4,d}^{(6)}, \hat{\mathcal{C}}_{4,u}^{(6)} \neq 0, \hat{\mathcal{C}}_{4,d}^{(6)} \neq 0\}$ (keeping all the other couplings zero). Here, the $\hat{\mathcal{C}}_{4,u}^{(6)} = \hat{\mathcal{C}}_{4,d}^{(6)}$ benchmark is special, since in this case the pion pole contribution vanishes. The small correction comes from the exchange of the next lightest pseudoscalar meson, $\eta$.

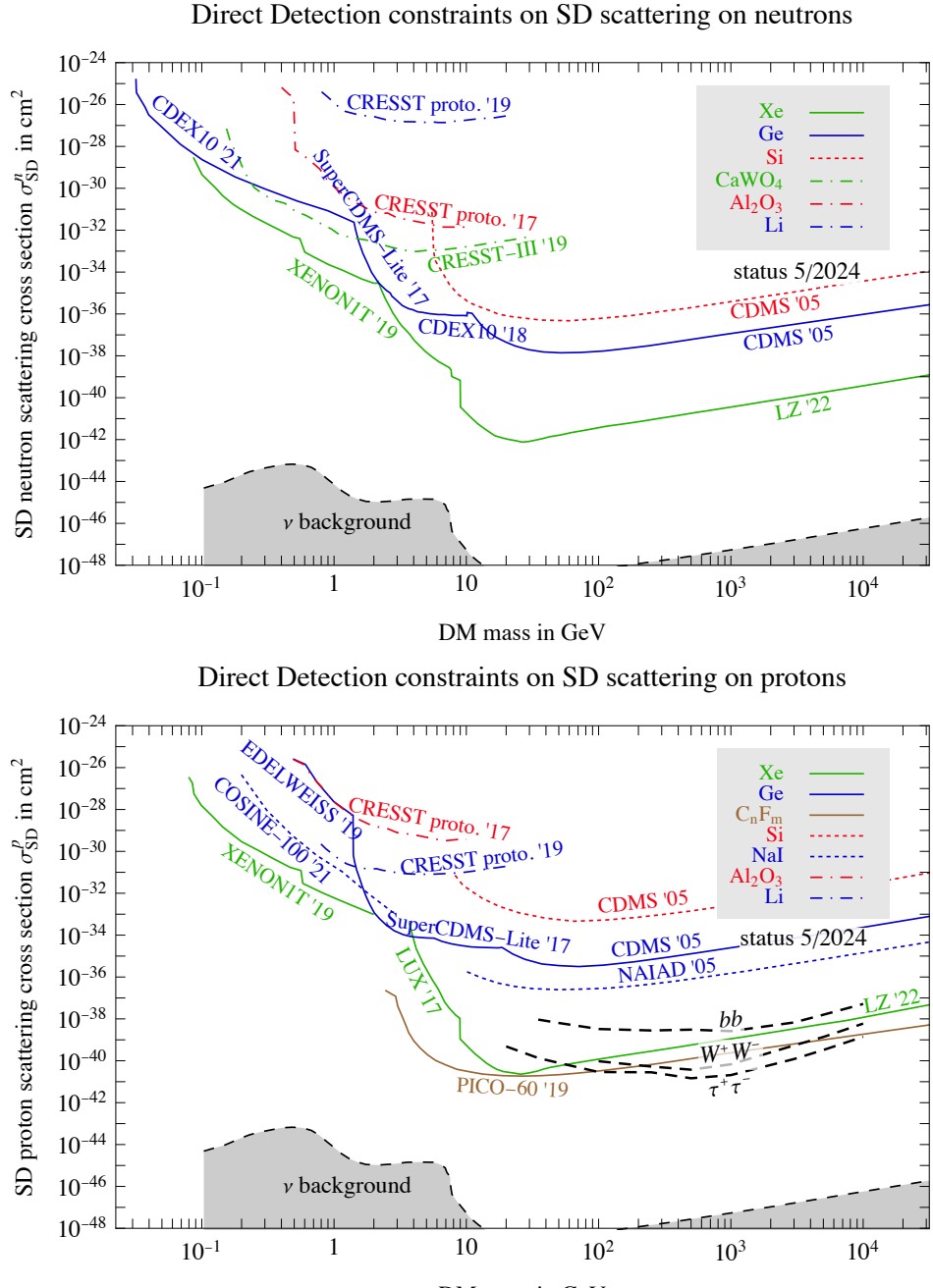

Figure 5.6: *As in fig. 5.5 (bottom), but for bounds on the **spin-dependent** direct detection cross sections on neutrons $\sigma_{SD}^n$ (top) and on protons $\sigma_{SD}^p$ (bottom) from underground detectors, and from* IceCube *[194] (black dashed lines, for the indicated DM annihilation modes that produce neutrinos from the Sun, as discussed in section 6.9).*

less than one for DM mass eigenstates that are mostly EW singlet with only a small admixture of the EW charged component.

Integrating out the $Z$ boson at tree level (see fig. 5.3a) gives the effective interaction between DM and the SM $\mathscr{L}_{\text{eff}} = -J^{\text{SM}} J^{\text{DM}} / M_Z^2$. This leads to SI scattering if DM is either a fermion with vector interactions ($g_{\text{DM}}^V \neq 0$), or if DM is a scalar. The non-relativistic coefficients of eq. (5.10) are given by

$$c_1^n = \left(\frac{g_2}{2c_W}\right)^2 \frac{g_{\text{DM}}}{M_Z^2} \approx 0.13 \frac{g_{\text{DM}}}{M_Z^2}, \qquad c_1^p = (4s_W^2 - 1)c_1^n \approx -0.01 \frac{g_{\text{DM}}}{M_Z^2}, \tag{5.25}$$

where $c_W \equiv \cos\theta_W$, $s_W \equiv \sin\theta_W$. Neglecting $|c_1^p| \ll |c_1^n|$ the SI scattering cross section is

$$\sigma_{\text{SI}} \approx \frac{1}{4}\sigma_{\text{SI}}^n = \frac{G_F^2 \mu_N^2}{2\pi} g_{\text{DM}}^2 \approx 7.4 \, 10^{-39} \, \text{cm}^2 \, \frac{\mu_N^2}{m_n^2} g_{\text{DM}}^2, \tag{5.26}$$

where $g_{\text{DM}} = g_{\text{DM}}^V (g_{\text{DM}}^S)$ for fermion (scalar) DM. Large DM couplings, $g_{\text{DM}} \sim \mathcal{O}(1)$, which one would obtain in the case of EW charged Dirac fermion DM, are excluded, see fig. 5.2. Direct detection data imply $g_{\text{DM}}^{V,S} \lesssim 2 \, 10^{-4}$ for electroweak scale DM masses, $M \sim \mathcal{O}(M_Z)$, while the bound becomes progressively less stringent for heavier or lighter DM, see fig. 5.5.

Fermionic DM with just the axial couplings only produces spin-dependent scattering. The vanishing of $g_{\text{DM}}^V = 0$ in eq. (5.24) is automatic for Majorana fermion DM, since for an anti-commuting Majorana fermions, $\bar{\chi}_i \gamma^\mu \chi_j = -\bar{\chi}_j \gamma^\mu \chi_i$, and thus Majorana fermion DM only has the axial coupling $g_{\text{DM}}^A$. Integrating out the $Z$ gives

$$\hat{\mathcal{C}}_{4,q}^{(6)} = \left(\frac{g_2}{2c_W}\right)^2 \frac{g_{\text{DM}}^A}{M_Z^2} T_3^q, \qquad \text{where} \qquad T_d^u = \frac{1}{2}, \quad T_3^d = -\frac{1}{2}, \tag{5.27}$$

with the non-relativistic coefficients for SD scattering given in eq. (5.21). This gives the same scattering cross section for DM scattering on protons and on neutrons, $\sigma_{\text{SD}}^p = \sigma_{\text{SD}}^n$, so that

$$\sigma_{\text{SD}} = J_{\text{DM}}(J_{\text{DM}} + 1)\frac{4G_F^2 \mu_N^2}{\pi}\left(g_{\text{DM}}^A\right)^2\left(\Delta u - \Delta d\right)^2. \tag{5.28}$$

The SD scattering is subject to a weaker direct detection bound, $\sigma_{\text{SD}} \lesssim 10^{-41} \, \text{cm}^2$, cf. fig. 5.6, and thus $g_{\text{DM}}^A \lesssim 10^{-2}$ for $M \sim M_Z$.

Small $g_{\text{DM}}$ couplings can arise if DM has a small mixing with a hypothetical state that carries a hypercharge $Y \sim 1$. Another possibility is that DM has a large coupling to a hypothetical heavy vector, $Z'$, and this has only a small mixing with the SM $Z$ boson. Integrating out the heavy $Z'$ gives an $\text{SU}(2)_L$ gauge-invariant higher dimension effective interaction, $\propto J_{\text{DM}}^\mu (H^\dagger D_\mu H)/M_{Z'}^2$, which after electroweak symmetry breaking gives a coupling between $Z$ and DM, $\mathscr{L}_{\text{int}} \propto J_{\text{DM}}^\mu Z_\mu v^2 / M_{Z'}^2$.

## Higgs-mediated DM interactions

The Higgs boson $h$ might couple to DM, $\mathscr{L}_{\text{int}} \supset -h J_h$, where[9]

$$J_h = J_h^{\text{SM}} + \begin{cases} \bar{\chi}(y_{\text{DM}} + i\gamma_5 \tilde{y}_{\text{DM}})\chi & \text{if DM is a fermion } \chi, \\ \lambda_{\text{DM}} v S^2/2 & \text{if DM is a real scalar } S, \\ \lambda_{\text{DM}} v S^* S & \text{if DM is a complex scalar } S, \end{cases} \tag{5.29}$$

---

[9]We work after electroweak symmetry breaking. The $\text{SU}(2)_L$-invariant generalization of the DM interaction to the Higgs is obtained by promoting $h$ to $H^\dagger H/(\sqrt{2}v)$, see table 9.1.

with $v \approx 174\,\text{GeV}$. Integrating out the Higgs boson (see fig. 5.3b) gives the effective interaction $\mathscr{L}_{\text{int}} \supset J_h^{\text{SM}} J_h^{\text{DM}} / M_h^2$. Scalar DM and fermionic DM with Yukawa interactions produce the spin-independent scattering with non-relativistic coefficients

$$c_1^n \simeq c_1^p = \frac{f_N}{\sqrt{2}} \frac{m_N}{M_h^2 v} \begin{cases} y_{\text{DM}} & \text{if DM is a Dirac fermion } \chi, \\ 2y_{\text{DM}} & \text{if DM is a Majorana fermion } \chi, \\ \lambda_{\text{DM}} v/2M & \text{if DM is a scalar } S, \end{cases} \tag{5.30}$$

where $f_N \approx 0.3$ is the Higgs coupling to nucleons,[10] resulting in the SI cross section

$$\sigma_{\text{SI}} = \frac{\mu_N^2}{2\pi} \left( \frac{f_N m_N}{v} \right)^2 \frac{1}{M_h^4} \begin{cases} y_{\text{DM}}^2 & \text{if DM is a Dirac fermion } \chi, \\ 4y_{\text{DM}}^2 & \text{if DM is a Majorana fermion } \chi, \\ (\lambda_{\text{DM}} v/2M)^2 & \text{if DM is a scalar } S. \end{cases} \tag{5.31}$$

Compared to the SI scattering induced by the tree level exchange of $Z$, eq. (5.25), the Higgs exchange induced scattering is suppressed by the Higgs couplings to matter, $f_N m_N / v$, and so the DM/Higgs couplings are less severely constrained. Numerically, $y_{\text{DM}} \lesssim 0.02$ for Dirac fermion DM with mass $M \sim \mathcal{O}(M_Z)$. Note that $y_{\text{DM}} \sim \mathcal{O}(1)$ has been excluded already for some time, see fig. 5.2. For heavier DM the bound on $\sigma_{\text{SI}}$ scales as $1/\sqrt{M}$, giving $y_{\text{DM}} \lesssim 0.1\sqrt{M/10\,\text{TeV}}$ for fermion DM, and $\lambda_{\text{DM}} \lesssim \left( M/2.0\ \text{TeV} \right)^{3/2}$ for scalar DM. The pseudo-scalar Yukawa coupling, $\tilde{y}_{\text{DM}}$, only gives spin-dependent scattering, which is further suppressed by the transferred momentum $q$, and is thus not subject to significant bounds.

The above cases for $\sigma_{\text{SI}}$ in eq. (5.31) can be shortened into a single result by noticing that the Higgs contribution to the DM mass and the DM-Higgs coupling are related, given that they both arise from the same Higgs doublet field, $\langle H \rangle \to v + h/\sqrt{2}$. Defining the Higgs-dependent DM mass $M(h) \gg m_N$, the tree level Higgs exchange induced direct-detection cross section is then for any DM spin given by,

$$\sigma_{\text{SI}} = \frac{g_{\text{DM}}^2}{2\pi v^2} \frac{f_N^2 m_N^4}{M_h^4}, \qquad \text{where} \quad g_{\text{DM}} = \frac{\partial M}{\partial h}, \tag{5.32}$$

and we approximated $\mu_N \simeq m_N$. For Dirac fermion DM, Majorana fermion DM, and scalar DM this reproduces the couplings in eq. (5.30), $g_{\text{DM}} = \{y_{\text{DM}}, 2y_{\text{DM}}, \lambda_{\text{DM}} v/2M\}$, respectively.

### $W$-mediated DM interactions

DM might be the neutral component of a weak multiplet with hypercharge $Y = 0$, in which case DM has a vanishing coupling to the $Z$ boson. The scattering of DM on a nucleon is then induced at one loop by the weak couplings of DM to the $W^\pm$ gauge bosons, whereby the virtual emission of a $W^\pm$ changes DM to a charged component of the electroweak multiplet (see fig. 5.3c). Integrating out the $W^\pm$ bosons gives a direct detection cross section

$$\sigma_{\text{SI}} \sim \alpha^6 m_N^2 / M_W^6, \tag{5.33}$$

which is around or below the present experimental bounds, see fig. 5.2. Theories of this type are discussed in more detail in section 9.3.4, see also fig. 9.5a.

### DM interactions mediated by a hypothetical dark photon

The interactions between DM and the SM can be mediated by light intermediate states. The difference with respect to the above results with the heavy mediators is that now even the $c_1^N$ in the non-relativistic

---

[10]It is given by $f_N = 2/9 + \sum_{u,d,s} 7\sigma_q^N/(9m_N)$, where $\sigma_q^N$ is the scalar sigma term for quark $q$, $\sigma_q^N \bar{u}_N u_N = \langle N | m_q \bar{q} q | N \rangle$, and have the values $\sigma_s^N \approx 40$ MeV, and $\sigma_{u,d}^N \approx 20 - 30$ MeV [182].

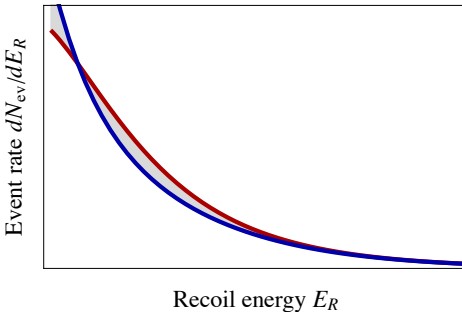
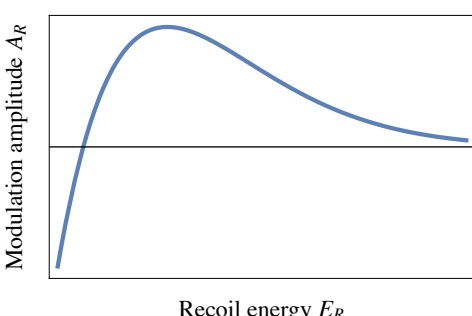

Figure 5.7: *Illustrative example of **DM-induced nuclear recoil rates** $dN_{\rm ev}/dE_R$ at $t =$June 2nd (red) and $t =$Dec. 2nd (blue dashed). The right handed plot shows the modulation amplitude $A_R(E_R)$.*

Lagrangian (5.10) are $q$ dependent, due to the propagator of the light mediator. An example of such a model is the dark photon, $A'_\mu$, a light vector boson that couples to the SM electromagnetic current, $J^\mu_{\rm em}$, and to DM, $\chi$,

$$\mathscr{L}_{\rm int} \supset -e\epsilon J^\mu_{\rm em} A'_\mu - g_D A'_\mu \bar{\chi}\gamma^\mu\chi, \tag{5.34}$$

where $\epsilon \ll 1$ (see section 9.4.1 for further details). The non-relativistic coefficients in (5.10) are given by

$$c^p_1 = -\frac{e\varepsilon g_D}{q^2 + m^2_{A'}}, \qquad c^n_1 = 0, \tag{5.35}$$

so that for $m_{A'}$ lighter than a few 10s MeV the $c^p_1$ becomes $q^2$ dependent (see fig. 5.1). This means that the SI cross section for scattering on nuclei has additional $q$ dependence,

$$\frac{d\sigma^A_{\rm SI}}{dE_R} = \frac{m_A}{2\pi v^2} Z^2 \big[c^p_1(q)\big]^2, \tag{5.36}$$

and therefore the bounds on DM scattering derived from direct detection under the assumption of a constant $c^N_1$ need to be reexamined. The kinetic mixing parameter is constrained by other searches to be $10^{-3} \lesssim \epsilon \lesssim 10^{-5}$ for $m_{A'} \gtrsim 10$ MeV, see fig. 7.5 below, while direct detection bounds require $\epsilon g_D \lesssim 10^{-10}$ for DM with electroweak scale mass. For lighter DM masses the bounds from scattering on nuclei become less stringent. For sub-GeV DM, the main constraints on dark photon mediated scattering are obtained from bounds on possible DM scatterings on electrons, for which the dark photon model is one of the main benchmark models (see also the discussion in section 5.2).

### 5.1.7   Annual modulation

The direct detection event rate $dR_{\rm ev}/dE_R$, eq. (5.4), contains an annual modulation induced by the Earth orbiting around the Sun [196], see figure 5.7. Indeed, as discussed in section 2.3.2, the DM gas is on average at rest with respect to the galactic frame[11], where DM particles have an average velocity $v_0 \approx \mathcal{O}(200)$km/s, see eq. (2.25). The Sun orbits the center of the Galaxy and moves through the DM gas with a speed of about $v_\odot \approx 233$ km/s, see eq. (2.29). This induces a 233 km/s DM wind in the frame of the solar system. In addition, the Earth orbits the Sun with velocity $V_\oplus \approx 29.8$ km/s, which induces an annual modulation in the DM wind speed observed on Earth, see also fig. 2.7. This affects the predicted DM scattering rate, $dR_{\rm ev}/dE_R$. The integrand in (5.4) — the DM flux, $v\, dn_{\rm DM}(\boldsymbol{v})$ — changes slightly throughout the year, because the relative velocity between the Earth and the DM halo changes. Because

---

[11]The possibility that the DM halo co-rotates with the disk is addressed in section 2.4.2.

the velocity integral has a fixed lower cut-off at the minimal velocity $v_{\min}$, this induces a modulation in $dR_{\rm ev}/dE_R$,

$$\frac{dR_{\rm ev}}{dE_R} = \overline{\frac{dR_{\rm ev}}{dE_R}} \left[ 1 + A_R(E_R) \cos\left(\omega(t - t_c)\right) \right], \tag{5.37}$$

where $\omega = 2\pi/{\rm yr}$ and we expanded to first order in $V_\oplus$. The annual modulation amplitude is of order $A_R \approx \mathcal{O}(V_\oplus/v_{0,\odot}) \approx 0.01 - 0.1$, with the maximal rate expected on $t_c = $ June 2, see eq. (2.34). The average DM speed and thereby nuclear recoil energy is higher at $t_c$ than at $t_c + {\rm yr}/2 = $ December 2nd. As a consequence, higher-energy events have a larger rate at $t_c$, while lower-energy events have a larger rate at $t_c + {\rm yr}/2$. If the velocity distribution $f_\oplus(\boldsymbol{v})$ is known, the value of $E_R$ at which $A_R(E_R)$ changes sign can be used to measure the DM mass $M$.

There is also a day-night modulation induced by the rotation of Earth around its axis. The **diurnal modulation** is maximal for a detector on the equator, where the surface speed due to Earth's rotation is $\approx 0.5\,{\rm km/s}$. Even for this maximal case the diurnal modulation is still smaller than the annual modulation and can usually be neglected. A possible exception are the diurnal modulations of average scattering directions, which could be used as a signal in the future directional DM detectors, cf. section 5.6.6. This is especially true for scattering of sub-GeV DM, with masses below 100 MeV, where the diurnal modulation can be significantly enhanced by anisotropies in the target material such as organic crystals, which are planned to be used in the detection of such sub-GeV DM. Similarly, annual modulation can be important to distinguish signal from backgrounds also for sub-MeV dark matter scattering on electrons, which we discuss in section 5.2.

### 5.1.8   Astrophysical uncertainties

The event rate for DM direct detection, eq. (5.4), depends on the product of the cross section times the local DM density, $\sigma_A \rho_\odot$,[12] and on the DM velocity distribution, $f_\oplus(\boldsymbol{v})$, eq. (2.31). Converting a bound on $\rho_\odot \sigma_A$ into a bound on $\sigma_A$ requires the determination of $\rho_\odot$ from astrophysical observations. While the local DM density has some uncertainty, see eq. (2.11), all the direct detection experiments assume a conventional value $\rho_\odot = 0.3\,{\rm GeV/cm^3}$ when interpreting their results, about 30% less than the present best value of $0.4\,{\rm GeV/cm^3}$ quoted in eq. (2.11) and used throughout this review. The commonly used value of $\rho_\odot$ facilitates a quick comparison between different experiments. A different $\rho_\odot$ implies a rescaling of all bounds on $\sigma_A$, as we did for the bounds shown in fig.s 5.5, 5.6, and 5.12.

The dependence of bounds on the local DM velocity distribution, $f_\oplus(\boldsymbol{v}) = f(\boldsymbol{v} + \boldsymbol{v}_\oplus(t))$, is less trivial. The astrophysical uncertainties in $f(\boldsymbol{v})$ can sometimes lead to large uncertainties. This is especially true for the $\sigma_{\rm SI,SD}(M)$ exclusion curves at low DM masses, controlled by the $E_{R\min}$ threshold, see fig. 5.4. In this parameter region the scatterings come from DM with velocities close to the Galactic escape velocity. For such velocities a large galactic variance of $f(\boldsymbol{v})$ is expected, as shown by the $N$ body simulations, see fig. 2.5. This translates to a significant uncertainty in the expected event rates in direct detection and complicates the comparison of direct detection bounds obtained using different nuclei. For simplicity this issue is often ignored and the bounds on DM scattering are plotted using the standard DM halo model, eq. (2.23).

If the dependence on $f_\oplus$ is important for the problem at hand, one can scan over a collection of reasonable DM velocity distributions, such as discussed in section 2.3. One can be even more conservative, however, and compare the results of direct detection experiments in a way that is independent of astrophysical uncertainties ('halo independent method'). The crucial insight is that the recoil energy $E_R$ is directly related to $v_{\min}$, see eq. (5.5). The distribution of events, $dR_{\rm ev}/dE_R$, can thus be translated into

---

[12]Strictly speaking it depends on DM density on Earth, $\rho_\oplus$. In the vast majority of cases this is the same as the DM density at the position of the solar system, $\rho_\oplus = \rho_\odot$. However, there are known exceptions, see, e.g., [197].

the $v_{\min}$ space. In the $v_{\min}$ space the properly normalised $dR_{\mathrm{ev}}/dE_R$ distributions from all experiments should look the same.

For instance, for SI scattering, where $d\sigma_A^{\mathrm{SI}}/dE_R \propto 1/v^2$, eq. (5.6), one can rewrite the scattering rate, eq. (5.4), as [198]

$$\frac{dR_{\mathrm{ev}}}{dE_R} = \tilde{\eta}(v_{\min})\frac{dH_A}{dE_R}, \tag{5.38}$$

where the velocity dependence is encoded entirely in the function

$$\tilde{\eta}(v_{\min}) = \frac{\rho_\oplus \sigma_{\mathrm{ref}}}{M} \int_{v_{\min}}^\infty d^3\boldsymbol{v}\, \frac{f_\oplus(\boldsymbol{v})}{v}, \tag{5.39}$$

where $v_{\min}$ is given in eq. (5.5). Here, $\sigma_{\mathrm{ref}}$ is a reference cross section, for which we take the average total cross section for DM scattering on a single proton and a single neutron, $\sigma_{\mathrm{ref}} = (\sigma_p^{\mathrm{SI}} + \sigma_n^{\mathrm{SI}})/2$ (other choices are equally viable). The remaining factor in eq. (5.38) does not depend on DM velocity profile at all

$$\frac{dH_A}{dE_R} = \frac{N_T}{m_T}v^2\frac{d\sigma_A^{\mathrm{SI}}}{dE_R} = \frac{m_A}{2m_T\mu_N^2}\frac{\sigma_{\mathrm{SI}}}{\sigma_{\mathrm{ref}}}A^2F^2(q). \tag{5.40}$$

In the last equality we used eq. (5.6). The function $dH_A/dE_R$ differs for different nuclei, because of changes in $m_A$, $\sigma_{\mathrm{SI}}$, $A$, and $F(q)$. On the other hand, the normalized scattering rate $\frac{dR_{\mathrm{ev}}}{dE_R}/\frac{dH_A}{dE_R}$ is the same for all nuclei, when expressed as a function of $v_{\min}$. Particle physics is encoded in $\sigma_{\mathrm{SI}}$ and $\sigma_{\mathrm{ref}}$, where $\sigma_{\mathrm{SI}}$ depends on the choice of the nuclear target, while $\sigma_{\mathrm{ref}}$ by definition does not. The ratio $\sigma_{\mathrm{SI}}/\sigma_{\mathrm{ref}}$ only depends on the ratio of DM couplings to protons and neutrons, $c_1^p/c_1^n$. Choosing this ratio, and fixing the DM mass $M$, completely determines $dH_A/dE_R$ for each nuclear target. Barring cancellations one has $\sigma_{\mathrm{SI}}/\sigma_{\mathrm{ref}} \sim \mathcal{O}(1)$, and thus the dependence of $dH_A/dE_R$ on the assumed particle physics is relatively mild. Furthermore, for not too light DM, $M \gg m_N$, the reduced mass $\mu_N \simeq m_N$ is almost completely independent of the value of $M$, and so is $dH_A/dE_R$.

Fixing the DM mass $M$ and 'particle physics', i.e., the ratio $c_1^p/c_1^n$, the experimental bound on $dR_{\mathrm{ev}}/dE_R$ implies a bound on $\tilde{\eta}(v_{\min})$, where $v_{\min} = (E_R m_A/2)^{1/2}/\mu_A$ is determined by the measured recoil energy, $E_R$. A nonzero signal, $dR_{\mathrm{ev}}/dE_R$, would similarly imply a measurement of $\tilde{\eta}(v_{\min})$. One can thus map all the bounds on (or measurements of) the scattering rates, $dR_{\mathrm{ev}}/dE_R$, into the exclusion lines (signal regions) in the $(v_{\min}, \tilde{\eta})$ plane. If a signal from one experiment, translated to a nonzero value of $\tilde{\eta}(v_{\min})$, is contradicted by a bound from another experiment, then the potential DM SI scattering signal is excluded irrespective of the DM velocity profile. Note that in this comparison the DM mass and particle physics couplings needed to be held fixed. To completely exclude the putative DM scattering signal one needs to marginalize over different values of $M$, and different ratios of particle physics couplings, $c_1^p/c_1^n$. The outlined statistical procedure is aided by the fact that $\tilde{\eta}(v_{\min})$ is a monotonically decreasing function. The approach has been extended to include annual modulation, as well as general DM couplings to nucleons [198].

### 5.1.9  Effective operators

If the interactions between dark and SM particles are mediated by particles heavier than the typical momentum exchange in direct detection, $q \lesssim 200$ MeV, it is convenient to integrate out the heavy mediators and end up with an Effective Field Theory (EFT) description of the interactions with DM (just like the electro-weak theory is approximated at low energies by Fermi four-fermion effective operators). As a simple example consider a scalar mediator, $\phi$, that couples to both the DM $\chi$ and quarks, with the interaction Lagrangian $\mathcal{L} \supset y_\chi\,\phi\bar{\chi}\chi + y_q\,\phi\bar{q}q$. The resulting DM/quark scattering amplitude $\mathcal{M}$ is given

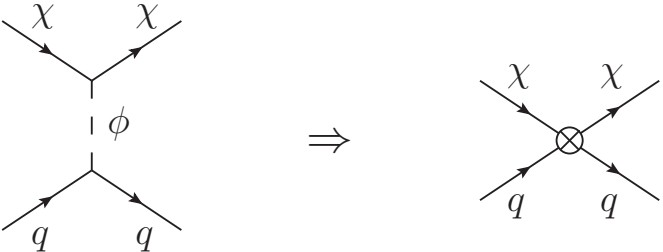

Figure 5.8: *Sample model where DM interacts with the SM via a scalar mediator $\phi$ that couples to quarks.*

by

$$i\mathcal{M} = -i\left(\bar{u}_\chi u_\chi\right)\frac{y_\chi y_q}{q_\mu q^\mu - m_\phi^2}\left(\bar{u}_q u_q\right) \overset{q \ll m_\phi}{\simeq} i\frac{y_\chi y_q}{m_\phi^2}\left(\bar{u}_\chi u_\chi\right)\left(\bar{u}_q u_q\right) + \mathcal{O}\left(\frac{q^2}{m_\phi^2}\right), \tag{5.41}$$

where we expanded in $q^2/m_\phi^2 \ll 1$ and kept the leading term only. DM scattering is then equally well described by the local interaction operator $\mathcal{C}\left(\bar{\chi}\chi\right)\left(\bar{q}q\right)$, where all the relevant physics of the mediator is encoded in the value of its coefficient, $\mathcal{C} = y_\chi y_q/m_\phi^2$. This EFT approach to direct detection is illustrated in fig. 5.8. It allows to be quite agnostic about the high-energy theory of DM: the only thing that matters for direct detection is the effective interaction.

Effective operators are conveniently ordered in terms of their dimensionality,

$$\mathscr{L}_{\mathrm{EFT}} = \sum_{a,d} \frac{\mathcal{C}_a^{(d)}}{\Lambda^{d-4}} \mathcal{O}_a^{(d)}. \tag{5.42}$$

Here the $\mathcal{C}_a^{(d)}$ are dimensionless coefficients encoding the UV couplings, while the scale $\Lambda$ can be identified with the mass of the mediators (in the above example $\Lambda = m_\phi$). The sums run over dimensions of the operators, $d = \{5, 6, 7, \ldots\}$, and labels $a$, denoting operators with differing field contents and Lorentz structures. Operators with lowest dimensions typically dominate the scattering rate, so that stopping at $d = 7$ provides a quite general and accurate approximation.

The EFT description of DM interactions has the following two main uses. First, it allows to relate the results of different direct detection experiments with minimal assumptions about the origin of DM interactions. For instance, if a single direct detection experiment observes a putative signal of DM, while all the other experiments only set limits, the immediate question would be: do the other experiments exclude such a DM signal? Using DM EFTs, one can answer such a question for all the models of DM for which the mediators are heavier than a few 100 MeV, such that a few coefficients $\mathcal{C}_a^{(d)}$ in eq. (5.42) describe all experiments.

Second, DM EFTs can be used to relate the processes involving DM that occur at very different energies, $\mu$. Concerning collider signals, the typical energy for DM production at the LHC (to be discussed in section 7.2.3) is set by the cuts on the visible final state particles and is typically around a TeV. Concerning indirect detection signals, annihilations of two non-relativistic DM particles release energy $2M$ into SM particles, so that the appropriate EFT that describes such indirect detection signals includes all particles lighter than DM. The structure of the EFT describing DM interactions depends on the typical energy (or momentum) exchange, the scale $\mu$. Instead of a single DM EFT there is a tower of different DM EFTs, depending on the scale $\mu$, as illustrated in fig. 5.9 for heavy mediators and electroweak scale DM mass (i.e. $\Lambda \gg v_{\mathrm{EW}} \sim M$).

At a particular scale $\mu$ the appropriate EFT includes the relevant propagating degrees of freedom.

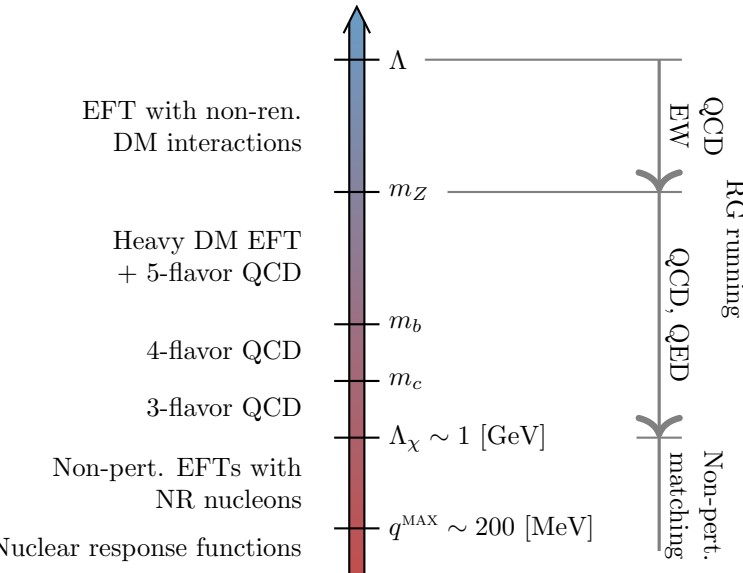

Figure 5.9: *The tower of effective field theories linking the UV scale $\Lambda$ to the scale of interactions between the nucleons and the DM. (From [199].)*

At the highest energy $\mu \sim \Lambda$ one needs the full theory of DM interactions. At $\mu \lesssim \Lambda$ the mediators can be integrated out, leading to an EFT with non-renormalizable interactions between the DM and SM particles. At $\mu \lesssim M_Z$ the top quark, the Higgs, and the $W, Z$ bosons can be integrated out and the DM interactions are therefore described by non-renormalizable operators in an EFT that contains only DM and the light SM particles: the photon, the gluons, the leptons, the $b$, $c$, $s$, $d$, $u$ quarks. At $\mu \sim m_b$ one integrates out the bottom quark, and at $\mu \sim m_c$ the charm quark. Finally, at $\mu \sim \mathcal{O}(1)$ GeV a non-perturbative matching to a chiral EFT with pions and nucleons is performed. This is then used to obtain the hadronic matrix elements for DM scattering on nuclei. Furthermore, DM can be conveniently described by a non-relativistic field at energies below its mass.

Quantum corrections can be important and need to be taken into account when obtaining the low-energy EFT from the high-scale physics at $\mu \sim \Lambda$. Fig. 5.10 illustrates three such examples of radiative corrections. For instance (left panel), if DM couples to the Higgs, integrating out the Higgs and the top quark at $\mu \simeq v_{\text{EW}}$ induces an effective coupling of DM to gluons. Even though this contribution is loop suppressed, it can be numerically important for direct detection scattering since nucleons have a large gluon content. Another example are the one loop exchanges of weak bosons, since these distinguish between left-handed and right-handed SM fields. Such radiative corrections can induce DM couplings to vector SM currents, even if in the UV theory the DM particle couples only to the axial SM currents (at tree level). Axial and vector currents have very different non-relativistic limits, and thus the induced mixing of operators can be numerically important. Finally, even if DM couples only to leptons at tree level, the radiative corrections such as the photon exchange in the right panel in fig. 5.10 induce DM couplings to quarks, and thus scattering of DM on nuclei.

In the rest of this section we focus on the DM EFT appropriate around the QCD scale $\mu \sim 2\,\text{GeV}$ and how such interactions of DM with quarks, gluons and photons induce scattering on nuclei. That is, we are interested in the last three steps in fig. 5.9: 3-flavor QCD, non-perturbative EFT with non-relativistic nucleons, and the nuclear response functions.

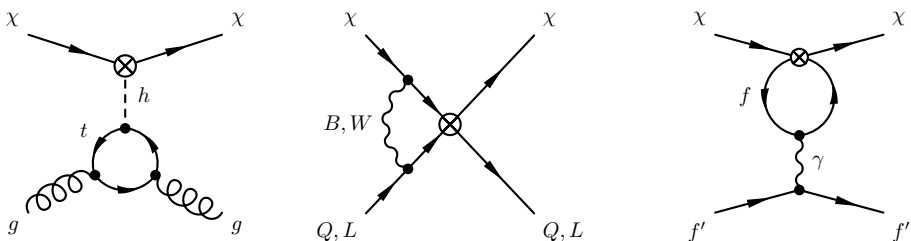

Figure 5.10: *Examples of one loop corrections that induce couplings to gluons (left), mix operators with different non-relativistic limits (middle), mix leptophilic and leptophobic interactions (right). (From [199].)*

## Quark level DM effective field theory

We first focus on the DM EFT with three quark flavors, which has as propagating degrees of freedom the light quarks, $u, d, s$, the gluons, photons, and SM leptons. For definiteness we set $\mu = 2\,\text{GeV}$, which is a conventional scale choice for lattice QCD determinations of the nuclear matrix elements. The form of the operators in the EFT Lagrangian in eq. (5.42) depends on whether DM is a scalar, a fermion or some higher spin particle. For instance, the complete basis for the EFT expansion up to dimension 7 contains 8 operators of dimension 6 for complex scalar DM. For a Dirac fermion DM there are two operators of dimension 5, four operators of dimension 6 and twenty-two operators of dimension 7, not counting fermion flavor multiplicities [200].

To gain further insight we focus on DM as a spin $1/2$ Dirac fermion $\chi$, and write the subset of operators, which are phenomenologically most interesting.

5) The two dimension-five operators are (in the notation of [200])

$$\mathcal{O}_1^{(5)} = \frac{e}{8\pi^2}(\bar{\chi}\sigma^{\mu\nu}\chi)F_{\mu\nu}, \qquad \mathcal{O}_2^{(5)} = \frac{e}{8\pi^2}(\bar{\chi}\sigma^{\mu\nu}i\gamma_5\chi)F_{\mu\nu}, \qquad (5.43)$$

where $F_{\mu\nu} = \partial_\mu A_\nu - \partial_\nu A_\mu$ is the electromagnetic field strength tensor. The magnetic dipole operator $\mathcal{O}_1^{(5)}$ is CP-even, while the electric dipole operator $\mathcal{O}_2^{(5)}$ is CP-odd. Both flip DM chirality. Due to gauge invariance these operators are generated by integrating out heavy mediators at loop level [201], and thereby include a loop suppression factor $e/8\pi^2$.

6) The dimension-six operators are

$$\mathcal{O}_{1,q}^{(6)} = (\bar{\chi}\gamma_\mu\chi)(\bar{q}\gamma^\mu q), \qquad\qquad \mathcal{O}_{2,q}^{(6)} = (\bar{\chi}\gamma_\mu\gamma_5\chi)(\bar{q}\gamma^\mu q), \qquad (5.44)$$

$$\mathcal{O}_{3,q}^{(6)} = (\bar{\chi}\gamma_\mu\chi)(\bar{q}\gamma^\mu\gamma_5 q), \qquad\qquad \mathcal{O}_{4,q}^{(6)} = (\bar{\chi}\gamma_\mu\gamma_5\chi)(\bar{q}\gamma^\mu\gamma_5 q). \qquad (5.45)$$

Here, $q = \{u, d, s\}$ denote the light quarks (we limit ourselves to flavor-conserving operators). Such dimension 6 operators are for instance generated by $Z$-mediated DM interactions discussed in section 5.1.6. While tree level $Z$ exchange generates all four types of coefficients, $\mathcal{C}_{i,q}^{(6)}$, the phenomenologically relevant ones are $\mathcal{C}_{1,q}^{(6)}$, since these induce SI scattering.[13] The other operators give SD scattering and can thus be neglected, if $\mathcal{C}_{i,q}^{(6)}$ is nonzero. The situation is different for Majorana fermion DM for which the $\mathcal{O}_{1,q}^{(6)}$ and $\mathcal{O}_{3,q}^{(6)}$ operators vanish, and thus SD scattering is the leading interaction.[14]

---

[13]Identifying $\Lambda = m_Z$ as the mass of the mediator gives $\mathcal{C}_{1,q}^{(6)} = g_{\text{DM}}\big(T_q^3 - 2s_W^2 Q_q\big)/2$, where $T_q^3 = 1/2$ is the weak isospin of $u$ quark and $T_q^3 = -1/2$ for $d, s$ quarks, and $Q_q$ the electromagnetic charge of quark $q$.

[14]For Majorana DM the definitions of the operators conventionally include an extra factor of $1/2$ in the four

7) A selection of (possibly phenomenologically most interesting) dimension-seven operators is

$$\mathcal{O}_1^{(7)} = \frac{\alpha_s}{12\pi}(\bar{\chi}\chi)G^{a\mu\nu}G^a_{\mu\nu}\,, \qquad\qquad \mathcal{O}_4^{(7)} = \frac{\alpha_s}{8\pi}(\bar{\chi}i\gamma_5\chi)G^{a\mu\nu}\widetilde{G}^a_{\mu\nu}\,, \qquad (5.46)$$

$$\mathcal{O}_{5,q}^{(7)} = m_q(\bar{\chi}\chi)(\bar{q}q)\,, \qquad\qquad \mathcal{O}_{9,q}^{(7)} = m_q(\bar{\chi}\sigma^{\mu\nu}\chi)(\bar{q}\sigma_{\mu\nu}q)\,, \qquad (5.47)$$

$$\mathcal{O}_{11}^{(7)} = \frac{\alpha}{12\pi}(\bar{\chi}\chi)\,F^{\mu\nu}F_{\mu\nu}\,, \qquad\qquad \mathcal{O}_{14}^{(7)} = \frac{\alpha}{8\pi}(\bar{\chi}i\gamma_5\chi)\,F^{\mu\nu}\tilde{F}_{\mu\nu}\,, \qquad (5.48)$$

where the labeling is as in [200], where also the remaining dimension 7 operators can be found. Here, $G^a_{\mu\nu}$ is the QCD field strength tensor, and $\widetilde{G}_{\mu\nu} = \frac{1}{2}\varepsilon_{\mu\nu\rho\sigma}G^{\rho\sigma}$ is its dual, and similarly for the electromagnetic field strength tensor $F_{\mu\nu}$ and its dual $\tilde{F}_{\mu\nu}$. The index $a = \{1,\ldots,8\}$ runs over color $SU(3)_c$ generators.

Dimension seven operators are encountered as the leading operators in many theories of DM. For instance, the Higgs-mediated DM interactions ($y_{DM} \neq 0$ in eq. (5.29)) generate the operator $\mathcal{O}_1^{(7)}$ at one loop after the Higgs and $t, b, c$ quarks are integrated out, see the left panel in fig. 5.10. The operators $\mathcal{O}_{5,q}^{(7)}$ arise from tree level Higgs exchanges, with the corresponding coefficients given by $\mathcal{C}_1^{(7)}/\Lambda^3 = -3\mathcal{C}_{5,q}^{(7)}/\Lambda^3 = 3y_{DM}/(\sqrt{2}vM_h^2)$. For this result, note that the proportionality of Higgs/quarks couplings to quark masses was already included in the definition of the $\mathcal{O}_{5,q}^{(7)}$ operators. Furthermore, the dependence on the heavy quark mass drops out in the result for the gluonic operator due to the required two chiral insertions in the loop cancelling the $1/m_q^2$ suppression from the loop propagators. The operators $\mathcal{O}_1^{(7)}$ and $\mathcal{O}_{5,q}^{(7)}$ all lead to SI scattering.

Pseudo-scalar mediators such as Goldstone bosons and axion-like (ALP) particles, on the other hand, couple to pseudo-scalar SM currents. Integrating out the ALP mediators coupling to gluons or photons generates the operators $\mathcal{O}_4^{(7)}$ and $\mathcal{O}_{14}^{(7)}$, respectively. If ALP couples to quarks, it would also generate the equivalent of $\mathcal{O}_{5,q}^{(7)}$ but with $i\gamma_5$ insertions in the scalar currents. All these operators lead to SD scattering.

The Rayleigh operators $\mathcal{O}_{11}^{(7)}, \mathcal{O}_{14}^{(7)}$, encode DM polarizability, which can be a sign of composite nature of DM or of (pseudo-)scalar mediators coupling to photons. Rayleigh operators are not the operators of lowest dimension that involve the DM and electromagnetic fields, because these arise already at dimension 5, see $\mathcal{O}_{1,2}^{(5)}$ in eq. (5.43). However, these magnetic and electric dipole moment operators vanish for Majorana fermion DM, so it may well be that the interactions to the SM only arise from DM polarizability, i.e., from the operators such as $\mathcal{O}_{11}^{(7)}, \mathcal{O}_{14}^{(7)}$. In fact, this would be a generic feature of models where DM is a Majorana fermion, that is not coupled to colored particles [203].

Eq.s (5.43) to (5.48) (plus the undisplayed dimension 7 operators) provide a complete basis of EFT operators of up to and including dimension 7. One could have chosen a different basis, with a different set of operators. Different choices of EFT bases describe the same physics: using equations of motion and performing integration per parts, one can convert these to the basis used here. For example, our basis does not include the DM anapole operator, $(\bar{\chi}\gamma_5\gamma^\mu\chi)\partial^\nu F_{\mu\nu}$, which is sometimes assumed to give dominant interactions for DM with higher dimension electromagnetic interactions (the so called *anapole DM* [204]). The electromagnetic equations of motion $\partial^\mu F_{\mu\nu} + \sum_f eQ_f\bar{f}\gamma_\nu f = 0$ allow to express the anapole operator in terms of the dimension 6 operators in eq. (5.44) and (5.45),

$$\bar{\chi}\gamma_5\gamma^\mu\chi\partial^\nu F_{\mu\nu} = -\sum_{q=u,d,s} eQ_q\mathcal{O}_{2,q}^{(6)} - \sum_{\ell=e,\mu,\tau} eQ_\ell\mathcal{O}_{2,\ell}^{(6)}. \qquad (5.49)$$

---

component notation, to compensate for the two possible contractions, simplifying the common notation for cross sections. See [202] for more details.

The EFT operators depend on DM spin. The EFT contains a smaller number of operators, if DM is a scalar $S$. We again list those relevant for the EFT at $\mu = 2$ GeV: the effective interactions only start at dimension six, with the full operator basis given by

$$\mathcal{O}_{1,f}^{(6)} = \left(S^* i \overset{\leftrightarrow}{\partial}_\mu S\right)(\bar{f}\gamma^\mu f), \qquad\qquad \mathcal{O}_{2,f}^{(6)} = \left(S^* i \overset{\leftrightarrow}{\partial}_\mu S\right)(\bar{f}\gamma^\mu \gamma_5 f), \tag{5.50}$$

$$\mathcal{O}_{3,f}^{(6)} = m_f(S^* S)(\bar{f}f), \qquad\qquad \mathcal{Q}_{4,f}^{(6)} = m_f(S^* S)(\bar{f}i\gamma_5 f), \tag{5.51}$$

$$\mathcal{O}_5^{(6)} = \frac{\alpha_{\rm s}}{12\pi}(S^* S)G^{a\mu\nu}G_{\mu\nu}^a, \qquad\qquad \mathcal{O}_6^{(6)} = \frac{\alpha_{\rm s}}{8\pi}(S^* S)G^{a\mu\nu}\widetilde{G}_{\mu\nu}^a, \tag{5.52}$$

$$\mathcal{O}_7^{(6)} = \frac{\alpha}{\pi}(S^* S)F^{\mu\nu}F_{\mu\nu}, \qquad\qquad \mathcal{O}_8^{(6)} = \frac{3\alpha}{\pi}(S^* S)F^{\mu\nu}\tilde{F}_{\mu\nu}, \tag{5.53}$$

where $S^* \overset{\leftrightarrow}{\partial}_\mu S \equiv S^*(\partial_\mu S) - (\partial_\mu S^*)S$. The operators $\mathcal{O}_4^{(6)}$, $\mathcal{O}_6^{(6)}$, and $\mathcal{O}_8^{(6)}$ are CP-odd, while all the others are CP-even. The above operators for scalar DM have a similar form to a subset of the effective operators for spin $1/2$ DM. The vector and axial vector currents operators are of dimension 6 both for scalar DM, eq. (5.50), and for fermion DM, eq.s (5.44), (5.45). The scalar operators in eq.s (5.51) to (5.52) correspond to the dimension 7 scalar operator $\mathcal{O}_{5,q}^{(7)}$ for fermion DM in eq. (5.44), along with the undisplayed operators with extra $i\gamma_5$ insertions. In each case there are less operators for scalar DM, because for fermionic DM one can insert $\gamma_5$ in the currents and thus have scalar and pseudo-scalar (or vector and axial-vector) currents on the DM side. In the non-relativistic limit such operators give as the leading effect the coupling to the DM spin for spin $1/2$ DM, which is then absent for spin 0 DM.

Another important consequence of scalar DM being spin 0 is that the spin $1/2$ DM can have a nonzero magnetic or electric moment, i.e., it is possible to write down the dimension 5 operators in eq. (5.43). There is nothing equivalent for scalar DM. For uncharged DM the first nonzero electromagnetic interaction are therefore dimension 6 polarizability operators, eq. (5.53).

## Heavy DM effective theory

The above DM EFT holds if DM is much lighter than the mediators, $M \ll \Lambda$. If this is not the case, the expansion in powers of $1/\Lambda$ becomes problematic: since $(M/\Lambda)^n$ is no longer small, higher and higher dimension operators give sizeable contributions. As a concrete example consider two operators: a dimension 6 vector-vector operator and a dimension 8 operator with extra derivatives acting on DM,

$$\frac{1}{\Lambda^2}(\bar{\chi}\gamma_\mu\chi)(\bar{q}\gamma^\mu q) \sim \frac{Mv_0^\mu}{\Lambda^2}(\bar{\chi}\chi)(\bar{q}\gamma^\mu q), \qquad \frac{1}{\Lambda^4}(\bar{\chi}\gamma_\mu\partial^2\chi)(\bar{q}\gamma^\mu q) \sim \frac{M^3 v_0^\mu}{\Lambda^4}(\bar{\chi}\chi)(\bar{q}\gamma^\mu q). \tag{5.54}$$

Above, we wrote the DM four-momentum as $p_\chi^\mu = Mv_0^\mu + k^\mu$, where $Mv_0^\mu$ with $v_0^\mu = (1,0,0,0)$ is the four-momentum of DM at rest in the laboratory frame, and $k^\mu \sim \mathcal{O}(Mv) \ll M$ is the remaining small momentum due to the relative motion of DM, with velocity $v \sim 10^{-3}$. In writing the r.h.s. expressions in eq. (5.54) we neglected corrections suppressed by powers of $k/M$. Eq. (5.54) shows that, after taking the non-relativistic DM wave functions, operators with dimension 6 and 8 contribute at the same level, if $M/\Lambda \sim \mathcal{O}(1)$. In the example in (5.54) they even correspond to the same non-relativistic operator. In order to fully describe DM scattering, the relativistic operators of arbitrarily high dimensions are then needed in $\mathscr{L}_{\rm EFT}$, a problematic situation, but they do collapse to a smaller set of non-relativistic operators.

The problem is that the total energy of DM, $E \simeq M$, is large, while its kinetic energy is small. The solution is a different EFT expansion that takes into account that DM is non-relativistic: an expansion in its small momentum, $k/M \sim k/\Lambda$, is again dominated by the few lowest orders. In effect, one integrates out the heavy DM mass but leaves in the DM particle. Technically, in the DM field $\chi$ we factor out the phase factor due to the propagation of the heavy DM mass, and split $\chi$ into particle and antiparticle

fields, by defining

$$\chi(x) = e^{-iMv_0 \cdot x}\big(\chi_v(x) + X_v(x)\big), \tag{5.55}$$

where

$$\chi_v(x) = e^{iMv_0 \cdot x}\frac{1 + \slashed{v}_0}{2}\chi(x), \qquad X_v(x) = e^{iMv_0 \cdot x}\frac{1 - \slashed{v}_0}{2}\chi(x). \tag{5.56}$$

The field $\chi_v$ describes the DM particle mode and is used to build the Heavy Dark Matter Effective Theory (HDMET)[15] [205], giving an expansion in $1/M$. It plays the same role as the heavy quark field in the Heavy Quark Effective Theory [206] that was first developed to deal with the non-perturbative nature of $b$ quark decays. HDMET can be used in two ways. If the complete UV model of DM is not known, HDMET still allows us to parameterize the scattering cross sections in terms of a few parameters, the coefficients of HDMET operators. If UV theory is known, matching onto HDMET is a technically useful step that simplifies calculations, for example by avoiding loop diagrams with very different scales. An example are precise predictions for scattering rates of pure wino and higgsino DM, mediated by other supersymmetric particles with comparable masses.

The remaining $x$ dependence in $\chi_v(x)$, eq. (5.56), is due to the soft momenta. Direct detection scattering changes the soft momentum of the DM by $q$ but does not change the DM velocity label $v_0$. The velocity label $v_0^\mu$ can be identified with either the incoming or the outgoing DM velocity four-vector, or any other velocity four-vector that is non-relativistically close to these two. The simplest choice is to identify $v_0^\mu$ with the lab frame velocity, $v_0^\mu = (1,0,0,0)$.

The "small-component" field $X_v$ in eq. (5.56) describes the antiparticle modes. To excite an antiparticle mode requires absorption of a hard momentum, of order $2M$. In constructing HDMET the anti-particle modes are integrated out. At tree-level this gives the following relation between $\chi_v$ and $\chi$ [205, 206]

$$\chi = e^{-iMv_0 \cdot x}\Big(1 + \frac{i\slashed{\partial}_\perp}{iv_0 \cdot \partial + 2M - i\epsilon}\Big)\chi_v, \tag{5.57}$$

where $\gamma_\perp^\mu = \gamma^\mu - v_0^\mu\slashed{v}$ and similarly $\partial_\perp^\mu = \partial^\mu - v_0^\mu v_0 \cdot \partial$. The second term in parentheses arises from integrating out the antiparticle mode $X_v$, and is explicitly of $\mathcal{O}(k/M)$, since $\partial_\perp$ acting on $\chi_v$ gives the soft momentum $\partial_\perp \sim \mathcal{O}(k)$. By substituting $\chi$ in terms of $\chi_v$ allows to build the HDMET Lagrangian

$$\mathscr{L}_{\text{HDMET}} = \bar{\chi}_v(iv \cdot \partial)\chi_v + \frac{\bar{\chi}_v(i\partial_\perp)^2\chi_v}{2M} + \cdots + \mathscr{L}_{\text{int}}. \tag{5.58}$$

The first term in eq. (5.58) is the leading-order HDMET Lagrangian for the free DM particle and contains no explicit dependence on $M$. It is also independent of DM spin, as there are no $\gamma^\mu$ factors. The second term in eq. (5.58) is the $\mathcal{O}(1/M)$ correction, and has the typical form of the non-relativistic kinetic energy, $k^2/2M$. The ellipses denote the corrections to the free particle Lagrangian of higher order in the $1/M$ expansion. Many of the operators in the interaction Lagrangian, $\mathscr{L}_{\text{int}}$, can be obtained from the operators for spin $1/2$ DM in the previous subsection, eq.s (5.43) to (5.48), adding to them also higher dimension operators, and replacing the relativistic DM currents with their leading nonzero expressions in the $1/M$ expansion, such as,

$$\begin{aligned} &\bar{\chi}\chi \to \bar{\chi}_v\chi_v, && \bar{\chi}i\gamma_5\chi \to \partial_\mu\big(\bar{\chi}_v S^\mu \chi_v\big)/M, && \bar{\chi}\sigma^{\mu\nu}\chi \to -2\epsilon^{\mu\nu\alpha\beta}v_{0,\alpha}\big(\bar{\chi}_v S_\beta \chi_v\big), \\ &\bar{\chi}\gamma^\mu\chi \to v_0^\mu\bar{\chi}_v\chi_v, && \bar{\chi}\gamma^\mu\gamma_5\chi \to 2\bar{\chi}_v S^\mu\chi_v, && \bar{\chi}\sigma^{\mu\nu}i\gamma_5\chi \to 2\bar{\chi}_v S^{[\mu}v_0^{\nu]}\chi_v, \end{aligned} \tag{5.59}$$

where $\epsilon^{\mu\nu\alpha\beta}$ is the totally antisymmetric Levi-Civita tensor, with $\epsilon^{0123} = 1$, and $S^{[\mu}v_0^{\nu]} = S^\mu v_0^\nu - S^\nu v_0^\mu$. In addition, there are also operators that are nonzero only for higher spin DM.

The $1/M$ expansion is just a formal way of taking the non-relativistic limit in a QFT. The expanded

---

[15]The name Heavy WIMP Effective Theory (HWET) is also used, especially if DM is part of an electroweak multiplet.

expressions in eq. (5.59) teach us something useful. The scalar and vector currents, $\bar{\chi}\chi$ and $\bar{\chi}\gamma^\mu\chi$, reduce to one unique DM number operator $\bar{\chi}_v\chi_v$, which counts the number of DM particles. Both currents thus result in SI interactions in the non-relativistic limit. The non-relativistic limit of $\bar{\chi}\gamma^\mu\gamma_5\chi$ involves the spin operator $S^\mu = \gamma_\perp^\mu\gamma_5/2$. The $S^\mu$ operator sandwiched between the DM non-relativistic states gives the DM spin: $\bar{u}S^\mu u = (0, \boldsymbol{S})\bar{u}u$ for DM at rest. The axial vector current therefore results in SD interactions. The pseudo-scalar current $\bar{\chi}i\gamma_5\chi$ also results in SD interactions, which, however, are also momentum suppressed. Finally, the tensor currents, i.e., the last two currents in eq. (5.59), result in unsuppressed SD interactions.

It is important to note that the procedure of replacing the relativistic currents with the $1/M$ expanded versions does not preserve the dimension of the operators. For instance, the non-relativistic limit of the dimension 7 operator $\mathcal{O}_{6,q}^{(7)} = m_q(\bar{\chi}i\gamma_5\chi)(\bar{q}q)$ gives an operator of one dimension higher, $\partial_\mu(\bar{\chi}_v S^\mu\chi_v)(\bar{q}q)$. This is suppressed by one power of soft momentum and thus more suppressed than one would have naively guessed just from the dimensionality of the relativistic operator. The opposite is also true. The two operators in the example of eq. (5.54) collapse to the same HDMEFT operator. More importantly, the following operators (of quite high dimension 8) have their dimensionality lowered by one when taking the non-relativistic limit due to the appearance of the large momentum $Mv_0^\mu$:

$$\mathcal{O}_{q(g)}^{(8)} = \left(\bar{\chi}\,i\overleftrightarrow{\partial}_\mu\gamma_\nu\chi\right)\mathcal{O}_{q(g)}^{\mu\nu} \to 2Mv_{0,\mu}v_{0,\nu}\left(\bar{\chi}_v\chi_v\right)\mathcal{O}_{q(g)}^{\mu\nu}. \tag{5.60}$$

The quark and gluon currents in eq. (5.60) are known as "twist-2" and given by

$$\mathcal{O}_q^{\mu\nu} = \frac{1}{2}\bar{q}\left(iD_-^{\{\mu}\gamma^{\nu\}} - \frac{g^{\mu\nu}}{4}i\slashed{D}_-\right)q, \qquad \mathcal{O}_g^{\mu\nu} = \frac{g^{\mu\nu}}{4}G^{a\rho\sigma}G_{\rho\sigma}^a - G^{a\mu\rho}G_\rho^{a\nu}. \tag{5.61}$$

In the previous subsection we assumed $M \ll \Lambda$ and considered the relativistic EFT up to dimension 7, and thereby ignored these dimension 8 operators in eq.s (5.43) to (5.48). These contributions become leading when DM and mediators have similar masses. HDMET makes sure that all such leading terms are kept.

Focusing on Spin-Independent scattering at $\mu = 2$ GeV, the tower of DMEFT operators collapses to just four leading HDMET operators:

$$\mathscr{L}_{\text{int}}^{\text{SI}} = \frac{1}{\Lambda_{\text{eff}}^3}\bar{\chi}_v\chi_v\left[c_{1q}^{(0)}m_q\bar{q}q + c_2^{(0)}\left(G_{\mu\nu}^a\right)^2 + v_{0,\mu}v_{0,\nu}\left(c_{1q}^{(2)}\mathcal{O}_q^{\mu\nu} + c_2^{(2)}\mathcal{O}_g^{\mu\nu}\right)\right]. \tag{5.62}$$

If $M \ll \Lambda$ the leading contributions to SI scattering are fully described by the two twist-0 coefficients $c_{1q}^{(0)}$, $c_2^{(0)}$, while for $M \sim \Lambda$ the twist-2 coefficients $c_{1q}^{(2)}, c_2^{(2)}$ are also required (but not the higher twists). An example where twist-2 contributions can be important is supersymmetric DM (section 10.1.2), if the DM neutralino is either part of the multiplet split only by electroweak terms (a higgsino or wino), or when squarks and/or gluinos are close in mass to the neutralino. The scale $\Lambda_{\text{eff}}$ is controlled by the mass splittings in the DM multiplet and by the mediators (the $W, Z, t, h$). For higgsino and wino DM these are both of electroweak size, thus $\Lambda_{\text{eff}} \sim v$ with $c_{1q}^{(0,2)}$ ($c_2^{(0,2)}$) one (two) loop suppressed.

## Nucleon level, non-relativistic

The final two steps in the tower of EFTs for direct detection in fig. 5.9 are the EFT that describes the interactions of DM with protons and neutrons, and the resulting scatterings on nuclei. Both DM and the nucleons inside nuclei are non-relativistic: the typical velocities are $v \sim 10^{-3}$ for DM and $v_N \sim 0.1$ for nucleons. This means that simple non-relativistic quantum mechanics can be used to describe the

interactions of DM with nucleons. The operators in the interaction Lagrangian

$$\mathscr{L}_{\mathrm{NR}} = \sum_{i,N} c_i^N(q^2)\mathcal{O}_i^N, \tag{5.63}$$

are constructed from Galilean-invariant quantities: the momentum exchange, $\boldsymbol{q} = \boldsymbol{k}_2 - \boldsymbol{k}_1$, the spin of DM and the nucleon, $\boldsymbol{S}$ and $\boldsymbol{S}_N$, as well as the relative velocity $\boldsymbol{v}_\perp = (\boldsymbol{p}_1 + \boldsymbol{p}_2)/2M - (\boldsymbol{k}_1 + \boldsymbol{k}_2)/2m_N$, where $\boldsymbol{p}_{1(2)}$ and $\boldsymbol{k}_{1(2)}$ are the incoming (outgoing) DM and nucleon three-momenta, respectively. The sum in eq. (5.63) runs over $N = p, n$ and over the non-relativistic operators, such as[16]

$$\mathcal{O}_1^N = 1_{\mathrm{DM}}1_N, \qquad\qquad\qquad \mathcal{O}_4^N = \boldsymbol{S}\cdot\boldsymbol{S}_N, \tag{5.64}$$

$$\mathcal{O}_5^N = \boldsymbol{S}\cdot\left(\boldsymbol{v}_\perp \times \frac{i\boldsymbol{q}}{m_N}\right)1_N, \qquad\qquad \mathcal{O}_6^N = \left(\boldsymbol{S}\cdot\frac{\boldsymbol{q}}{m_N}\right)\left(\boldsymbol{S}_N\cdot\frac{\boldsymbol{q}}{m_N}\right). \tag{5.65}$$

Above, we list only the non-relativistic operators that will be needed below. The non-relativistic operators can be grouped in terms of how many derivatives they contain, i.e., the number of insertions of $q/m_N \sim \mathcal{O}(0.1)$ and $v_\perp \sim \mathcal{O}(0.1)$. There are in total 14 operators with up to two derivatives (times 2, one set for $N = p$ and another for $N = n$), with the full list given in Anand et al. (2013) [192]. The expectation is that the operators with more derivatives are more suppressed. However, we already saw around eq. (5.21) that the $q^2$ dependence of the non-relativistic coefficients $c_i^N(q^2)$ can compensate the derivative suppression in the operators. Is it then enough to include non-relativistic operators up to two derivatives? Should one include also operators with four, six,..., derivatives? Luckily, in the limit of heavy mediators (heavier than a few 100 MeV), all the $q^2$ dependence in $c_a^N(q^2)$ is due to known strong and electromagnetic interactions, and can be kept track of. A non-perturbative matching that we explain below leads to the leading order expressions

$$c_1^N = -\frac{\alpha}{2\pi M}Q_N\hat{\mathcal{C}}_1^{(5)} + \sum_q\left(F_1^{q/N}\hat{\mathcal{C}}_{1,q}^{(6)} + F_S^{q/N}\hat{\mathcal{C}}_{5,q}^{(7)}\right) + F_G^N\hat{\mathcal{C}}_1^{(7)} + \cdots, \tag{5.66a}$$

$$c_4^N = -\frac{2\alpha}{\pi}\frac{\hat{\mu}_N}{m_N}\hat{\mathcal{C}}_1^{(5)} - 4\sum_q F_A^{q/N}\hat{\mathcal{C}}_{4,q}^{(6)} + \cdots, \tag{5.66b}$$

$$c_5^N = \frac{2\alpha Q_N m_N}{\pi\boldsymbol{q}^2}\hat{\mathcal{C}}_1^{(5)}, \tag{5.66c}$$

$$c_6^N = \frac{2\alpha}{\pi\boldsymbol{q}^2}\hat{\mu}_N m_N\hat{\mathcal{C}}_1^{(5)} + \sum_q F_{P'}^{q/N}\hat{\mathcal{C}}_{4,q}^{(6)} + \cdots, \tag{5.66d}$$

where we shortened the coefficients in eq. (5.42) as $\hat{\mathcal{C}}_1^{(d)} \equiv \mathcal{C}_1^{(d)}/\Lambda^{d-4}$. The form factors $F_a^{q/N}$ and $F_G^N$ are evaluated at $q^2 = 0$, except for $F_{P'}^{q/N}$, which is given by eq. (5.72) below. The ellipses denote the contributions from the other DM EFT operators in $\mathscr{L}_{\mathrm{EFT}}$, eq. (5.42), that we did not touch upon above. The complete matching expressions can be found in [182].

In order to obtain eq. (5.66) we need a non-perturbative matching between $\mathscr{L}_{\mathrm{EFT}}$ in eq. (5.42), the DM EFT with quarks and gluons,[17] and operators in eq.s (5.43) to (5.48), onto the non-relativistic interaction Lagrangian $\mathscr{L}_{\mathrm{NR}}$, eq. (5.63). Since the momentum exchange between DM and nucleus is relatively small, $q \lesssim 200$ MeV, the quarks and gluons remain confined inside hadrons during the scattering. The appropriate degrees of freedom are non-relativistic protons and neutrons and the light $\pi, \eta$ mesons.

---

[16] Our definition of $\boldsymbol{q}$ follows [207] (and thus differs from Anand et al. (2013) in [192] by a minus sign). The definitions of the operators coincide between Anand et al. (2013) [192] and [207]. Since the system is non-relativistic the four momentum $q^\mu$ is dominated by the spatial component, $q^\mu \simeq (0, \boldsymbol{q})$ and thus $q^\mu q_\mu \simeq -\boldsymbol{q}^2 = -q^2$.

[17]We focus on the non-relativistic DM EFT valid for DM lighter than mediators; a similar matching can be performed for HDMET.

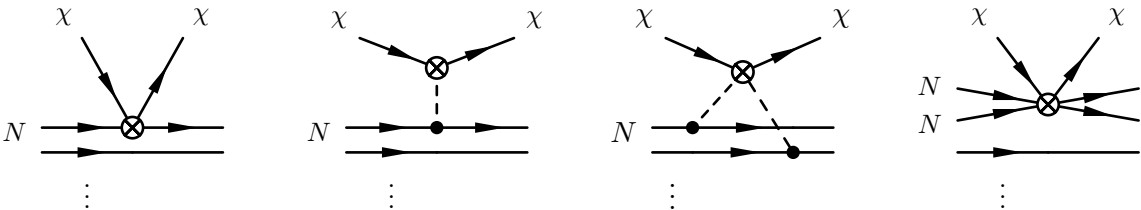

Figure 5.11: *The first and second panel from the left show the diagrams for DM-nucleus scattering that are of leading order in chiral counting. The long distance (third diagram) and short distance (fourth diagram) two nucleon interactions are examples of higher order correction in chiral counting. The effective DM–nucleon and DM–meson interactions is denoted by a circle, the dashed lines denote mesons, and the dots represent the remaining $A-2$ nucleon lines. Adapted from [207] (©IOP Publishing, reproduced with permission).*

For such small momenta exchanges, the strong nuclear force exhibits a special symmetry structure, a spontaneous breaking of a global $\mathrm{SU}(3)_L \otimes \mathrm{SU}(3)_R$ flavor symmetry, due to which the interactions between light mesons are derivatively suppressed. The various contributions can then be organized using Chiral Perturbation Theory (ChPT) [208]. Each contribution to the scattering amplitude scales as $M \sim (p/4\pi f)^\nu$, where $p \sim m_\pi \sim q$ is the typical momentum exchange, while $4\pi f = 1.2$ GeV is the cut-off of the ChPT effective theory. The suppression power $\nu$ depends on the form of the diagram and what types of vertices it contains — for instance, more derivatives in the vertices gives a higher suppression, as do more loops. ChPT can be extended to describe the interactions with a single non-relativistic nucleon, giving the so called heavy baryon ChPT [209], or the interactions between nucleons, giving the chiral EFT for nuclear forces [210].

There are several general lessons that one can draw from the above chiral expansion [182, 207, 211]. Perhaps most importantly, the chirally-leading contributions to DM scattering involve only scattering on a single nucleon. Such contributions are of two types. If DM couples to a nucleon through a point interaction (the first diagram in fig. 5.11) the resulting non-relativistic coefficient $c_i^N$ in (5.63) is $q$-independent. If instead DM interacts with a nucleon by emitting an off-shell light meson, $\pi^0$ or $\eta$, or a photon (the second diagram in fig. 5.11), this results in light meson or photon poles, $c_i^N \propto 1/(q^2 + m_\pi^2)$ and $c_i^N \propto 1/\boldsymbol{q}^2$. An explicit example of the former was encountered in eq. (5.21). The pole counteracts $\mathcal{O}(q^2)$ suppressions. Therefore, the proper description of leading non-relativistic DM interactions in $\mathscr{L}_{\mathrm{NR}}$, eq. (5.63), requires us to keep operators with up to two derivatives (but not with more derivatives).

For a number of DM EFT operators the hadronization is quite simple, and results, at leading order

in the chiral expansion, in just one non-relativistic operator. For instance,[18]

$$\mathcal{O}_{1,q}^{(6)} \to F_1^{q/N} \mathcal{O}_1^N, \qquad \mathcal{O}_1^{(7)} \to F_G^N \mathcal{O}_1^N, \qquad \mathcal{O}_{5,q}^{(7)} \to F_S^{q/N} \mathcal{O}_1^N, \qquad \mathcal{O}_{9,q}^{(7)} \to 8 F_{T,0}^{q/N} \mathcal{O}_4^N, \tag{5.68}$$

where all the form factors are evaluated at zero momentum transfer, $q^2 = 0$, resulting in constant non-relativistic coefficients, $c_{1,4}^N$. The non-relativistic reductions of vector-vector DM interactions ($\mathcal{O}_{1,q}^{(6)}$, eq. (5.44)), as well as scalar DM interactions with gluons and quarks ($\mathcal{O}_1^{(7)}$, $\mathcal{O}_{5,q}^{(7)}$ in eq.s (5.46), (5.46)), all lead to SI scattering (they generate $c_1^N$). These were used in section 5.1.6 to obtain the SI scattering cross sections for Higgs-mediated, eq. (5.31), and for $Z$-mediated DM interactions, eq. (5.25). The tensor-tensor operator $\mathcal{O}_{9,q}^{(7)}$, eq. (5.47), on the other hand, leads to spin-dependent scattering (it generates $c_4^N$, see also eq. (5.22)).

On the other hand, the chirally-leading hadronization is quite complicated for the magnetic moment operator $\mathcal{O}_1^{(5)}$ in eq. (5.43) and for the axial-axial operator $\mathcal{O}_{4,q}^{(6)}$ in eq. (5.45):

$$\mathcal{O}_1^{(5)} \to -\frac{\alpha}{2\pi} Q_N \left( \frac{1}{M} \mathcal{O}_1^N - 4 \frac{m_N}{q^2} \mathcal{O}_5^N \right) - \frac{2\alpha}{\pi} \frac{\hat{\mu}_N}{m_N} \left( \mathcal{O}_4^N - \frac{m_N^2}{q^2} \mathcal{O}_6^N \right), \tag{5.69}$$

$$\mathcal{O}_{4,q}^{(6)} \to -4 F_A^{q/N} \mathcal{O}_4^N + F_{P'}^{q/N} \mathcal{O}_6^N, \tag{5.70}$$

where $Q_N$ ($\hat{\mu}_N$) is the nucleon's electric charge (magnetic moment, $\hat{\mu}_p \approx 2.793$, $\hat{\mu}_n \approx -1.913$), while $F_A^{q/N}(q^2), F_{P'}^{q/N}(q^2)$ are the form factors for the axial current

$$\langle N' | \bar{q} \gamma^\mu \gamma_5 q | N \rangle = \bar{u}'_N \left[ F_A^{q/N}(q^2) \gamma^\mu \gamma_5 + \frac{1}{2m_N} F_{P'}^{q/N}(q^2) \gamma_5 q^\mu \right] u_N. \tag{5.71}$$

The form factors parametrise the response of a nucleon to an axial-vector probe $\bar{q}\gamma^\mu\gamma_5 q$ that gives a four-momentum $q^\mu$ kick to the initial nucleon in a state $|N\rangle \equiv |N(k_1)\rangle$, transforming it into the final nucleon state $\langle N'| \equiv \langle N(k_2)|$. The expressions in eq.s (5.69), (5.70) illustrate that a single relativistic operator can result in several non-relativistic operators $\mathcal{O}_i^N$, and that the coefficients $c_i^N$ multiplying them depend on $q^2$. This $q^2$ dependence is known. Let us first focus on the case of DM magnetic moment, eq. (5.69). Tree-level photon exchange between DM and a nucleon gives rise to terms with $1/q^2$ poles (multiplying $\mathcal{O}_{5,6}^N$ operators), as expected for a propagating photon. It also leads to $q^2$-independent contact terms (multiplying $\mathcal{O}_{1,4}^N$ operators) because the magnetic moment interaction in eq. (5.43) contains derivatives that can completely cancel the photon pole. The photon poles in eq. (5.69) then only multiply the operators with two derivatives, i.e., the two operators $\mathcal{O}_{5,6}^N$ that are $\mathcal{O}(q^2)$. All four terms in eq. (5.69) are thus of the same parametric order. Which contribution is the most important numerically, the SI contribution from $\mathcal{O}_1^N$ or the SD contributions from $\mathcal{O}_{4,5,6}^M$, depends on the type of nucleus and on the

---

[18]We define the form factors following [182]:

$$\langle N' | \{ \bar{q}\gamma^\mu q, m_q \bar{q}q, \frac{\alpha_s}{8\pi} G^{a\mu\nu} \tilde{G}_{\mu\nu}^a, m_q \bar{q}\sigma^{\mu\nu} q \} | N \rangle = \bar{u}'_N \{ F_1^{q/N} \gamma^\mu + \cdots, F_S^{q/N}, F_G^N, F_{T,0}^{q/N} \sigma^{\mu\nu} + \cdots \} u_N, \tag{5.67}$$

where the ellipses denote the form factors that, for the operators we are considering, contribute only at subleading orders in chiral expansion. At zero momentum exchange the vector currents count the number of valence quarks in the nucleon. Hence, the normalization of the Dirac form factors for the proton is $F_1^{u/p}(0) = 2, F_1^{d/p}(0) = 1, F_1^{s/p}(0) = 0$. The scalar form factors $F_S^{q/N}$ evaluated at $q^2 = 0$ are conventionally referred to as the nuclear sigma terms, $F_S^{q/N}(0) = \sigma_q^N \sim \mathcal{O}(10s \text{ MeV})$. Another common notation is $\sigma_q^N = m_N f_{Tq}^N$. The trace of the stress-energy tensor gives for the gluonic form factor $F_G^N(0) = -\frac{2}{27}(m_N - \sum_q \sigma_q^N)$. The tensor form factor at zero recoil is given by $F_{T,0}^{q/p}(0) = m_q A_{T,10}^{q/N}(0) = m_q g_T^q$, where the values of the tensor charges $g_T^q(\mu)$ are listed below eq. (5.23). The arrows in (5.68) denote the replacements of operators to be done in (5.42), which then give $\mathcal{L}_{\rm NR}$ in (5.63).

DM mass $M$.

Finally, we turn to the hadronization of the axial-axial operator, eq. (5.70). The $q^2$ dependence in the form factors $F_{A,P'}^{q/N}(q^2)$ leads to the $q^2$ dependence in the non-relativistic coefficients $c_{4,6}^N$ in $\mathscr{L}_{\text{NR}}$, eq. (5.63). We only need the chirally-leading behaviour, which amounts to expanding in small $q^2/m_N^2$

$$F_A^{q/N}(q^2) = \Delta q_N + \cdots, \qquad F_{P'}^{q/N}(q^2) = \frac{m_N^2}{m_\pi^2 + q^2}a_{q/N} + \frac{m_N^2}{m_\eta^2 + q^2}\tilde{a}_{q/N} + \cdots. \tag{5.72}$$

Both the value of $F_A^{q/N}(0) = \Delta q_N$ and the residua of the $\pi$ and $\eta$ poles, $a_{u/p} = -a_{d/p} = 2g_A$, $a_{s/p} = 0$, $\tilde{a}_{u/p} = \tilde{a}_{d/p} = -\tilde{a}_{s/p}/2 = 2(\Delta u_p + \Delta d_p - 2\Delta s_p)/3$, are related to the axial charges of the proton, $\Delta u \equiv \Delta u_p = 0.897(27)$, $\Delta d \equiv \Delta d_p = -0.376(27)$, $\Delta s \equiv \Delta s_p = -0.031(5)$, where $g_A = \Delta u - \Delta d = 1.2723(23)$ [182]. The expressions for neutrons are obtained by making the replacements $p \to n, u \leftrightarrow d$ (no change is implied for $g_A$). The resulting explicit forms of $c_{4,6}^N$, neglecting the $\eta$ pole, are given in eq. (5.21). The leading ChPT expression for $c_4^N$ is $q^2$ independent, i.e., the $\mathcal{O}_4^N$ term corresponds to the first diagram in fig. 5.11. The $c_6^N$ contains the light meson poles, and is thus given by the second diagram in fig. 5.11. The pion pole enhancements compensates the $\mathcal{O}(q^2)$ suppression in the $\mathcal{O}_6^N$ operators, and thus the two terms in eq. (5.70) are of parametrically similar size.

This completes our discussion of leading nonperturbative matching. The matching expressions in eq.s (5.66) get corrected at higher orders in chiral expansion. Going to higher orders requires inclusion of DM interactions with two nucleons. An example is the third diagram in fig. 5.11, where DM scatters off a pion exchanged between two nucleons. Such 'long distance' contributions are expected to be most important for axial-vector–vector, scalar–scalar and pseudo-scalar–scalar forms of DM/SM interactions, for which they enter as $\mathcal{O}(q)$ corrections in chiral counting. In contrast, the short distance two-nucleon contributions, 4th diagram in fig. 5.11, are nominally only an $\mathcal{O}(q^3)$ correction. However, non-perturbative renormalizations can change these scaling. Resumming ladder diagrams with long distance insertions may require counter-terms (short distance) contributions that are formally of higher order. This is understood for $np, nn, pp$ scattering to be a consequence of a new scale appearing in the low energy phenomenology — the large scattering length and the existence of bound states, such as deuteron [210]. It can also be understood as a change of order due to divergence of two-nucleon wave function at the origin [212]. Such effects were found to be important for neutrino-less double-beta decays, where the long distance two body interaction is due to a tree level exchange of a light neutrino [213]. Two-body current contributions were included for scalar DM interactions in the mean field approximation in [214] and found to be numerically relevant only when the chirally leading terms have large cancellations. For most applications the chirally leading calculations thus suffice, especially in view of sizable systematic errors in the evaluation of some of the nuclear response functions that we discuss next.

### Nuclear response functions

Restricting the discussion to the chirally-leading order, the cross section for the DM scattering on a nucleus is obtained by taking the matrix element of DM interactions with single nucleon currents in the non-relativistic effective Lagrangian $\mathscr{L}_{\text{NR}}$, eq. (5.63),

$$\left|\mathscr{A}\right|_{\text{nucleus,NR}}^2 = \sum_{i,j} \sum_{N,N'} c_i^N c_j^{N'*} \langle A|\mathcal{O}_i^N|A\rangle\langle A|\mathcal{O}_j^{N'}|A\rangle, \tag{5.73}$$

where the sum runs over $N, N' = n, p$, and all the non-relativistic operators with up to two derivatives, $i, j = 1, \ldots, 14$ (see eq.s (5.64), (5.65) and Anand et al. (2013) [192]). The high-energy particle physics is encoded in the non-relativistic coefficients $c_i^N$. The products of nuclear matrix elements $\langle A|\mathcal{O}_i^N|A\rangle\langle A|\mathcal{O}_j^{N'}|A\rangle$, on the other hand, depend only on nuclear physics, and can be calculated once and

for all. Once averaged over nuclear spin (we consider only scatterings on unpolarized nuclei), this results in eight nuclear response functions needed for the description of DM scattering on nucleus at leading order in chiral expansion. The nuclear response functions depend on $q = |\boldsymbol{q}|$ and have the approximate scalings

$$W_M \sim \mathcal{O}(A^2), \qquad W_{\Phi''M} \sim \mathcal{O}(A), \qquad W_{\Sigma'}, W_{\Sigma''}, W_\Delta, W_{\Delta\Sigma'}, W_{\tilde{\Phi}'}, W_{\Phi''} \sim \mathcal{O}(10^{-2}) - \mathcal{O}(1). \quad (5.74)$$

The $W_{\Sigma'}, W_{\Sigma''}, W_\Delta$, and $W_{\Delta\Sigma'}$ response functions depend strongly on the detailed properties of nuclei, for instance, whether or not these have an unpaired nucleon in the outer shell. Here:

* $W_{\Sigma',\Sigma''}$ encode the spin content of the nucleus discussed in section 5.1.5.

* $W_\Delta$ encodes the average angular momentum in the nucleus, and $W_{\Delta\Sigma'}$ the interference of the two.

* $W_{\tilde{\Phi}'}$ and $W_{\Phi''}$ encode the size of spin-orbit coupling in the nucleus. They can thus differ drastically between different isotopes of the same element.

* $W_M$ encodes the coherent scattering enhancement, $\mathcal{O}(A^2)$, where $A$ is the atomic mass number.

* $W_{\Phi''M}$ gives the interference between spin-orbit coupling operator and the coherent scattering.

The coherence is achieved in the long-wavelength limit, $q \to 0$, where DM scatters coherently on the whole nucleus, for instance, due to the $\mathcal{O}_1^N$ contact interaction (see also the discussion in section 5.1.4).

The DM/nucleus scattering cross section is given by [192],

$$\frac{d\sigma^A}{dE_R} = \frac{2m_A}{(2J_A + 1)v^2} \sum_{\tau,\tau'} \left[ R_M^{\tau\tau'} W_M^{\tau\tau'} + R_{\Sigma''}^{\tau\tau'} W_{\Sigma''}^{\tau\tau'} + R_{\Sigma'}^{\tau\tau'} W_{\Sigma'}^{\tau\tau'} + \frac{q^2}{m_N^2} \left( R_\Delta^{\tau\tau'} W_\Delta^{\tau\tau'} + \right. \right.$$

$$\left. \left. + R_{\Delta\Sigma'}^{\tau\tau'} W_{\Delta\Sigma''}^{\tau\tau'} + R_{\tilde{\Phi}'}^{\tau\tau'} W_{\tilde{\Phi}'}^{\tau\tau'} + R_{\Phi''}^{\tau\tau'} W_{\Phi''}^{\tau\tau'} + R_{\Phi''M}^{\tau\tau'} W_{\Phi''M}^{\tau\tau'} \right) \right]. \quad (5.75)$$

Here, $E_R$ is the recoil energy of the nucleus, $m_A$ the mass of the nucleus, $J_A$ its spin, and $v$ the initial DM velocity in the lab frame. The kinematical factors contain the $c_i^N$ coefficients,

$$R_M^{\tau\tau'} = c_1^\tau c_1^{\tau'} + \frac{1}{4} \frac{q^2}{m_N^2} v_T^2 c_5^\tau c_5^{\tau'} + \cdots, \quad (5.76a)$$

$$R_{\Sigma''}^{\tau\tau'} = \frac{1}{16} \left( c_4^\tau c_4^{\tau'} + \frac{q^2}{m_N^2} (c_4^\tau c_6^{\tau'} + c_6^\tau c_4^{\tau'}) + \frac{q^4}{m_N^4} c_6^\tau c_6^{\tau'} \right), \quad (5.76b)$$

$$R_{\Sigma'}^{\tau\tau'} = \frac{1}{16} c_4^\tau c_4^{\tau'} + \cdots, \qquad R_\Delta^{\tau\tau'} = \frac{1}{4} \frac{q^2}{m_N^2} c_5^\tau c_5^{\tau'} + \cdots, \qquad R_{\Delta\Sigma'}^{\tau\tau'} = \frac{1}{4} c_5^\tau c_4^{\tau'} + \cdots, \quad (5.76c)$$

where $\boldsymbol{v}_T = (\boldsymbol{p}_1 + \boldsymbol{p}_2)/2M - (\boldsymbol{k}_1 + \boldsymbol{k}_2)/2m_A$, with $\boldsymbol{p}_{1(2)}$ and $\boldsymbol{k}_{1(2)}$ the incoming (outgoing) DM and nuclear three-momenta, respectively. Note that for heavy nuclei such as xenon $\boldsymbol{v}_T \sim v \sim \mathcal{O}(10^{-3})$ is much smaller than the equivalent perpendicular velocity $v_\perp \sim \mathcal{O}(0.1)$ for the scattering on nucleons, see also the discussion surrounding eq. (5.65). For light nuclei, on the other hand, $\boldsymbol{v}_T \sim v_\perp$. The sum in eq. (5.75) is over isospin, $\tau, \tau' = 0, 1$, so that

$$c_i^{0(1)} = \frac{1}{2} (c_i^p \pm c_i^n). \quad (5.77)$$

In eq.s (5.76) we set $J = 1/2$ for simplicity and only displayed the dependence on the coefficients for the non-relativistic operators we focused on in the previous subsection, eq.s (5.64), (5.65). The complete expressions for $R_i^{\tau\tau'}$ can be found in Anand et al. (2013) in [192].

The form of the kinematical functions (5.76a)-(5.76c) agrees with the chiral counting we discussed in the previous subsection. The coefficients $c_{5,6}$, enhanced by a $1/q^2$ pole, come suppressed by either $q^2/m_N^2$ or $v_T^2$, and thus lead to parametrically similar contributions as the $c_1$ and $c_4$ coefficients. The new ingredient that was not captured by the chiral counting is the possibility of coherent enhancement of the nuclear response functions, eq. (5.74). Both $\mathcal{O}_1^N$ and $\mathcal{O}_5^N$ operators contain the nuclear number operator $1_N$ and thus exhibit coherent enhancement in eq. (5.76a). However, for $\mathcal{O}_5^N$ all the nucleons need to scatter in the same direction, converting the individual $\boldsymbol{v}_\perp$ factor to their sum, resulting in a much smaller $v_T$ suppression, effectively negating the coherent enhancement.

The precision of the cross section prediction for the DM scattering on nuclei in eq. (5.75) depends in large part on how well the nuclear response functions can be predicted. While ab initio nuclear calculations are possible for light nuclei, and for some more symmetric not-so-light nuclei such as $^{40}$Ca, this is not the case for heavy nuclei such as Xe, W, where various approximations are needed. Typically, calculations are done using the shell model with different choices for nuclear potentials. Anand et al. (2013) in [192] provided shell model calculations of response function $W_i$ for most nuclei used in direct detection experiments. Alternative calculations for spin independent response functions are available in [215] (though not using the $W_i$ notation, but also including two body current corrections). The systematic errors of these calculations are not easy to estimate, though some guidance is provided by comparing the results of different groups, when these are available. For $W_M$ the precision of a few percent is possible, while for other $W_i$ the errors are typically expected to be much larger.

### Modifications for light mediators

Throughout this section we assumed that the mediators are heavier than a few GeV, so that the use of DM EFT Lagrangian, eq. (5.42), is warranted. What happens, if mediators are lighter? Since light mediators are necessarily color neutral (they do not feel the strong force), the required changes to the above formalism are rather straightforward. The point interaction between DM and SM fields gets replaced by long range interactions. That is, the Wilson coefficients in (5.42) become $q^2$ dependent. Apart from this modification all the results carry through unchanged. A concrete example of this type is the dark photon mediator. The resulting $q^2$-dependent $c_1^N$ coefficient, eq. (5.35), can be used directly in the expression for the differential cross section for DM scattering on nuclei, eq. (5.75).

### 5.1.10 Numerical codes

A number of computer codes compute DM direct detection scattering rates. `DirectDM` [202] in combination with `DMFormFactor` [192] calculates the scattering cross sections for general DM interactions with the SM, based on the EFT approach discussed in section 5.1.9. This includes the SI and SD scattering, but also allows for more general interactions, starting either from the theory of DM interactions with quarks, gluons and photons, or from a non-relativistic description of DM interactions with nucleons. Another package with a similar scope is `ChiralEFT4DM` [215]. `MadDM` [216] calculates the SI and SD scattering rates starting from a Lagrangian for DM interactions with the SM. It is a plug-in for the very popular automatized matrix element generator `MadGraph5_aMCNLO` [217], and as such can utilize other popular tools such as `FeynRules` [218] that takes as input the DM interaction Lagrangian. The code `micrOMEGAs` [191] calculates the SI and SD scattering rates for a general set of DM models for which these are the leading scattering rates, and also provides the recasting of limits from experimental results to the space of DM couplings [219]. Such recasting can also be achieved using `DarkBit` [220].

## 5.2 Scattering on electrons

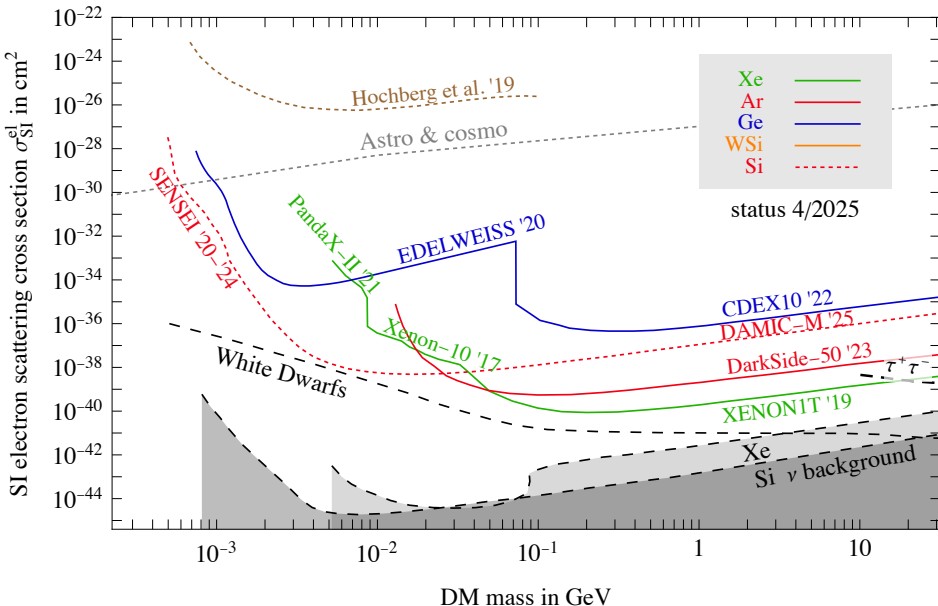

Figure 5.12: *The same as in fig. 5.5 (bottom), but for bounds from underground detectors on the spin-independent direct detection cross section for **scattering on electrons** $\sigma_{\rm SI}^{\rm el}$ in the limit of heavy mediators, see eq. (5.83). The estimate for neutrino backgrounds on Xe and Si targets is taken from Carew et al. (2024) in [221] (see also section 5.6.5). The White Dwarf bounds assume annihilating DM, see section 6.10.3. Black dashed line with the $\tau^+\tau^-$ label indicates the bound due to possible $DM + DM \to \tau^+\tau^-$ annihilation that would produce neutrinos from the Sun, see section 6.9. The Hochberg et al. '19 bound is obtained using superconducting nanowires, see section 5.2.3 and [222]. The astrophysical and cosmological bound is taken from Buen-Abad et al. (2022) [187], see section 5.1.2.*

Scattering of DM on electrons is most important for probing sub-GeV DM, for which the mass of the target — the electron — is closer to the mass of DM and thus the energy transfer is most efficient. Lighter DM implies smaller kinetic energy, and thus smaller deposited energy and momenta exchanges in the scattering events. The deposited energy can even be comparable to the energy excitations of the collective modes in the target materials, such as plasmons or magnons, which then need to be taken into account. This is especially true for the absorption of a very light, $\mathcal{O}(10\,\text{eV})$, DM in the material in which case also the kinematics matches with such collective mode excitations. The resulting current constraints on SI DM scattering on electrons are given in fig. 5.12.

To be able to describe the effects of scattering on bound electrons we need to extend the formalism of section 5.1, used for DM scattering on nuclei, such that it applies to the case of scattering of DM on bound electrons.[19] Starting with the Fermi's Golden rule the event rate per unit target mass, $R_{\rm ev} = dN_{\rm ev}/dt \times 1/m_T$, for DM scattering in a target with a volume $V$ and mass $m_T$ is given by [223, 224]

$$R_{\rm ev} = \frac{1}{\rho_T} \int dn_{\rm DM}(\boldsymbol{v}) \frac{V d^3\boldsymbol{p}_2}{(2\pi)^3} \sum_f \left| \langle f, \boldsymbol{p}_2 | H_{\rm int} | i, \boldsymbol{p}_1 \rangle \right|^2 2\pi \delta(E_f - E_i + E_2 - E_1), \qquad (5.78)$$

where $\rho_T = m_T/V$ is the mass density of the target material, $\boldsymbol{p}_{1(2)}$ and $E_{1(2)}$ are the incoming (outgoing) DM three momenta and energies, $H_{\rm int}$ the non-relativistic Hamiltonian describing DM–electron interac-

---

[19]We follow the review [223], where many more details can be found.

tions, $|i\rangle$ and $|f\rangle$ are the initial and final state of the detector with energies $E_i$, $E_f$, respectively, while $n_{\text{DM}}(\boldsymbol{v})$ is the volume density of DM particles with velocity $\boldsymbol{v} = \boldsymbol{p}_1/M$, see eq. (5.4). The wave functions are unit normalized, $\langle i|i\rangle = \langle f|f\rangle = \langle \boldsymbol{p}_i|\boldsymbol{p}_i\rangle = 1$. The factor $V d^3\boldsymbol{p}_2/(2\pi)^3$ gives the density of states for the outgoing DM plane wave. The event rate is also proportional to the number of DM particles, $V \int dn_{\text{DM}}(\boldsymbol{v})$. Normalizing to the target mass, $m_T$, then results in the appearance of the $\int dn_{\text{DM}}(\boldsymbol{v})/\rho_T$ factor in eq. (5.78).

Eq. (5.78) generalizes the expression for DM scattering on a single nucleus, eq. (5.4), by not requiring that the scattering is $2 \to 2$ and not imposing the corresponding kinematic constraints, except for energy conservation. Eq. (5.78) therefore allows for DM scattering on many electrons at the same time (on collective modes, the quasi-particles). The simplification in the DM/electron scattering problem is that for most applications both DM and electrons are non-relativistic so that one can use the usual non-relativistic quantum mechanics to evaluate the matrix element in eq. (5.78). However, the electrons are inside the medium, in a strongly interacting regime, so that the problem is far from a simple one. Luckily, one can use the tools developed for condensed matter problems to make headway.

The non-relativistic DM/electron interaction Hamiltonian takes a form of a product of DM and electron currents

$$H_{\text{int}} = \int \frac{d^3\boldsymbol{q}}{(2\pi)^3} e^{i\boldsymbol{q}\cdot(\boldsymbol{r}_{\text{el}}-\boldsymbol{r}_{\text{DM}})} \sum_a c_a^{\text{el}}(q)\, \mathcal{O}_{\text{DM}}^a \times \mathcal{O}_{\text{el}}^a, \tag{5.79}$$

where the various operators are the same as for the non-relativistic DM interactions with the nucleons, eq. (5.63), but with nucleons replaced by electrons: $m_N \to m_e$ and $\boldsymbol{S}_N \to \boldsymbol{S}_e$. The matrix elements therefore factorize into the DM and electron (target) parts, $\langle f, \boldsymbol{p}_2|H_{\text{int}}|i, \boldsymbol{p}_1\rangle \sim c_{\text{el}}^a \langle \boldsymbol{p}_2|\mathcal{O}_{\text{DM}}^a|\boldsymbol{p}_1\rangle \langle f|\mathcal{O}_{\text{el}}^a|i\rangle$. The DM matrix elements $\langle \boldsymbol{p}_2|\mathcal{O}_{\text{DM}}^a|\boldsymbol{p}_1\rangle$ and the values of the coefficients $c_{\text{el}}^a(q)$ change from one DM model to another. In particular, for heavy mediators $c_{\text{el}}^a$ are constant, while for light mediators they are $q$-dependent due to the propagator of the light mediator, signalling that in this case $H_{\text{int}}$ is nonlocal. For instance, a tree level exchange of a dark photon induces the operator $\mathcal{O}_1^{\text{el}} = 1_{\text{DM}}1_{\text{el}}$ with a coefficient (see also section 9.4.1)

$$c_1^{\text{el}}(q) = -\frac{\varepsilon e g_{\text{D}}}{q^2 + m_{A'}^2}, \tag{5.80}$$

where $\varepsilon$ is the kinetic mixing parameter, $\mathscr{L} \supset \varepsilon F'_{\mu\nu}F^{\mu\nu}/2$, which induces the coupling of strength $\varepsilon e$ between electrons and the dark photon, while $g_D$ is its coupling to DM. If the dark photon mass $m_{A'}$ is comparable to the typical momentum exchange, the $q$ dependence in $c_1^{\text{el}}(q)$ cannot be neglected.

Unlike the $c_a^{\text{el}}(q)$ coefficients, which change model to model, the response of the target material, encoded in $|\langle f|\mathcal{O}_{\text{el}}^a|i\rangle|^2$ and in similar interference terms between different operators, can in principle be calculated once and for all. This has been performed for a number of operators and materials, such as the isolated atom and crystals, in [225]. However, by far the main focus of the research so far has been on spin-independent DM–electron scattering, partially because the most popular sub-GeV DM physics models lead to SI scattering, but also because the response to SI scattering can be related to the leading behaviour of material under electromagnetic probes and is thus better explored.

From now on we thus limit the discussion to the SI interaction

$$H_{\text{int}}^{\text{SI}} = \int \frac{d^3\boldsymbol{q}}{(2\pi)^3} e^{i\boldsymbol{q}\cdot(\boldsymbol{r}_{\text{el}}-\boldsymbol{r}_{\text{DM}})} c_1^{\text{el}}(q) 1_{\text{DM}} 1_{\text{el}}, \tag{5.81}$$

for which the scattering rate can be written as

$$R_{\text{ev}}^{\text{SI}} = \frac{1}{\rho_T} \int dn_{\text{DM}}(\boldsymbol{v}) \int \frac{d^3\boldsymbol{q}}{(2\pi)^3} d\omega \frac{\pi\bar{\sigma}(q)}{\mu_e^2} \delta(\omega + E_2 - E_1) S(\boldsymbol{q}, \omega), \tag{5.82}$$

where $\mu_e$ is the reduced mass of the electron/DM system, $\mu_e = m_e M/(m_e + M)$. The quantity

$$\bar{\sigma}(q) = \frac{\mu_e^2}{\pi}[c_1^{\text{el}}(q)]^2, \tag{5.83}$$

reduces to the SI cross section for scattering of DM on free electrons, $\sigma_{\text{SI}}^{\text{el}}$, in the limit of a heavy mediator, $c_1^{\text{el}}(q) \to c_1^{\text{el}}(0)$. In general, however, it is $q$-dependent, and encodes the strength of the DM/electron interactions. In the literature, a common notation is to introduce the fiducial cross section $\sigma_T \equiv \bar{\sigma}(q_0)$ at a fixed momentum $q_0$, typically the inverse Bohr radius, $q_0 = 1/a_0 = \alpha m_e \simeq 3.7$ keV, and write $\bar{\sigma}(q) = \sigma_T F_{\text{DM}}^2(q)$, where $F_{\text{DM}}(q) \equiv c_1^{\text{el}}(q)/c_1^{\text{el}}(q_0)$ is the *DM form factor*. This is somewhat of a misnomer since the $q$ dependence of $F_{\text{DM}}(q)$ does not signal that DM is a composite object but rather comes from propagation of light mediators. For heavy mediators $F_{\text{DM}} \to 1$, while for light mediators $F_{\text{DM}} \to (q_0/q)^2$.

The *dynamic structure factor*,

$$S(\boldsymbol{q}, \omega) = \frac{2\pi}{V} \sum_f \left| \langle f | \sum_k e^{i\boldsymbol{q}\cdot\boldsymbol{r}_k} | i \rangle \right|^2 \delta(E_f - E_i - \omega), \tag{5.84}$$

gives the response of the material to the probe $e^{i\boldsymbol{q}\cdot\boldsymbol{r}_{\text{el}}} 1_{\text{el}}$, resulting in the sum $\sum_k$ over all the electrons in the target. The variable $\omega$ in eq. (5.82) gives the energy deposited in the material during the scattering event. In deriving eq. (5.82) from (5.78) the conservation of DM three-momenta, $\boldsymbol{q} = \boldsymbol{p}_1 - \boldsymbol{p}_2$ was used to convert the $d^3\boldsymbol{p}_2$ integral to an integral over $d^3\boldsymbol{q}$. However, the derivation of eq. (5.82) did not assume the conservation of the three-momentum in the target. In fact, since the crystal lattice breaks translation invariance, the eigenstates of the target are in general not eigenstates of momentum.

For scattering on a dilute non-relativistic free electron gas at negligible temperature (i.e., for electrons essentially at rest) the structure function is given by

$$S(\boldsymbol{q}, \omega) = 2\pi n_e \, \delta\left(\omega - \frac{q^2}{2m_e}\right), \tag{5.85}$$

where $n_e$ is the electron number density. The delta function in eq. (5.85) ensures that the deposited energy $\omega$ satisfies the non-relativistic dispersion relation. The limit of scattering on free electrons is approached for large momenta exchanges, $q \gg 1/a_0 = \alpha m_e = 3.73$ keV (the inverse of Bohr radius $a_0$ represents the typical extent of the electron wave-function) and for energy deposits $\omega \gg \alpha^2 m_e$ (the typical atomic electron binding energy). That is, for $q \gtrsim \mathcal{O}(\text{keV})$ and $\omega \gtrsim \mathcal{O}(10\,\text{eV})$ the response of the material to the DM/electron interactions peaks around the free electron dispersion $\omega = q^2/2m_e$, shown schematically as the green band in fig. 5.13 (left). The momenta exchanges and energy deposits of this size are achieved only for scattering of heavy enough DM, with mass above $\mathcal{O}(\text{GeV})$. For lighter DM masses it is important to take into account that the electrons are bound. First of all, many of the materials have energy gaps so that a minimal DM kinetic energy is required to be able to excite electrons, which then implies that there is a lower limit on the DM masses that can be probed by a given detector material. For very light sub-GeV DM, i.e., with a sub-MeV mass, the collective effects also become important.

There is a completely general kinematical limit as to how big the energy deposit $\omega$ in the DM scattering event can be. From energy conservation we have

$$\omega(\boldsymbol{q}) = E_1 - E_2 = \frac{1}{2}Mv^2 - \frac{1}{2M}(M\boldsymbol{v} - \boldsymbol{q})^2 = \boldsymbol{q}\cdot\boldsymbol{v} - \frac{q^2}{2M} \leq qv - \frac{q^2}{2M}, \tag{5.86}$$

so that the deposited energy, $\omega \leq Mv^2/2$, lies below a DM mass dependent parabola, shown with blue in fig. 5.13 (left panel, note that the log scale is used). The deposited energy is smaller for lighter DM. The kinematically allowed $(\omega, q)$ region, where the free electron gas approximation can be used, is reached

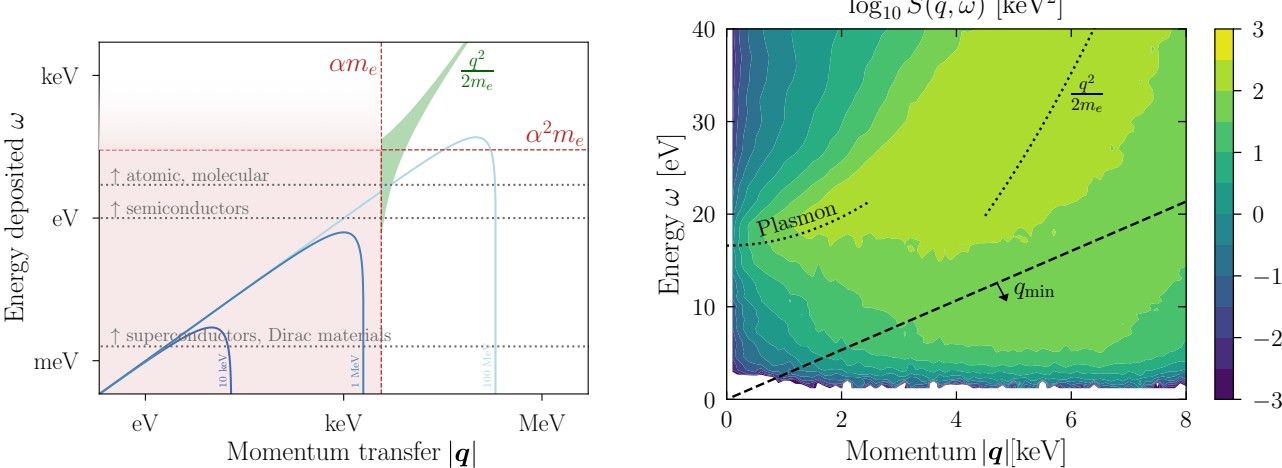

Figure 5.13: **Left**: *schematical kinematic regions for DM scattering on electrons with q the momentum transfer and ω the energy deposit. The green band shows the typical response of the material close to the region where the free electron gas approximation becomes reasonable. The blue regions show the kinematically allowed energy deposits, ω, as a function of the momentum exchange, $|\boldsymbol{q}|$, for DM masses $M = 10\,\mathrm{keV}, 1\,\mathrm{MeV}, 100\,\mathrm{MeV}$ and initial DM velocity $v = 10^{-3}$. The dashed red lines show the typical scale of the wave-function spread ($q \sim 1/a_0 = \alpha m_e$) and the energy ($\alpha^2 m_e$) of bound electrons. The fact that the electrons are bound inside the material is typically important in the red shaded region. There is a lower bound on the required deposited energy ω due to energy gaps with typical values for different systems as indicated on the grey dotted lines. For small ω and q also many body effects can become important. **Right**: the density plot of the dynamical structure factor $S(q,\omega)$ for* Si. *From [223] (©IOP Publishing, reproduced with permission).*

only for $M$ in the GeV regime and above. To have a good sensitivity to DM we want the structure function $S(\boldsymbol{q},\omega)$ to be sizeable in the kinematically allowed region, which means that different materials may be most suitable to be used as targets for various DM masses. In particular, searching for sub-GeV to sub-MeV DM requires materials with small band gaps, as low as meV, such that DM scattering can still cause electronic response, see fig. 5.13 (left). This is sometimes called *kinematic matching*. Depositing energy $\omega$ through DM scattering requires a minimal exchanged momentum

$$q_{\min} = \frac{\omega}{v_{\max}} \simeq 3.7\,\mathrm{keV}\left(\frac{\omega}{10\,\mathrm{eV}}\right)\left(\frac{800\,\mathrm{km/s}}{v_{\max}}\right), \tag{5.87}$$

where $v_{\max}$ is the largest DM velocity in the Earth's frame, $v_{\max} \simeq v_\oplus + v_{\mathrm{esc}} \simeq 800\,\mathrm{km/s}$, see section 2.3. The required momenta exchanges can be large compared to the typical momenta in the condensed matter systems, implying suppressed sensitivity to DM (for an example of such suppression see the discussion following eq. (5.90) below).

## 5.2.1 Ionization

The ionization energies in atomic systems are controlled by the Rydberg constant, $R_\infty = 13.6$ eV, i.e., the ionization energy of hydrogen, and are $\mathcal{O}(10\,\mathrm{eV})$ for the outer shell electrons in noble gases such as xenon and argon, and about $\mathcal{O}(5\,\mathrm{eV})$ for excitation gaps in organic molecules. This means that one can probe DM masses from tens of MeV to GeV using such systems as targets, since DM carries enough kinetic

energy to ionize the atom or the molecule. However, even in that case there is a suppression associated with the fact that depositing $\mathcal{O}(10\,\mathrm{eV})$ in a system for typical DM velocities requires minimal momentum exchanges that is larger than the inverse size of the bound electron wave functions, $q_{\min} > 1/a_0$. The DM scattering cross sections are thus suppressed either by powers of $1/(q_{\min}a_0)$, due to suppression of electron wave functions at high momenta [226], or require DM from tails of velocity distributions, see eq. (5.87). Using eq. (5.84) the dynamic structure function for the ionization of atoms is

$$S(\boldsymbol{q},\omega) = 2\pi n_{\mathrm{atom}} \int \frac{d^3 \boldsymbol{k}_e}{(2\pi)^3} \delta(E_f - E_i - \omega)\big|f_{0\to \boldsymbol{k}_e}(\boldsymbol{q})\big|^2, \tag{5.88}$$

with $k_e = |\boldsymbol{k}_e| = [2m_e(\omega - E_B)]^{1/2}$ the momentum of the outgoing electron, $n_{\mathrm{atom}}$ the number density of atoms, and $E_B$ the binding energy. The *atomic form factor*, $f_{0\to \boldsymbol{k}_e}(\boldsymbol{q})$, for the electron transition from the ground state to the continuum state with momentum $\boldsymbol{k}_e$, is given by the weighted overlaps of the initial and final wave functions. For the hydrogen atom it is

$$f_{0\to \boldsymbol{k}_e}(\boldsymbol{q}) = \int d^3 \boldsymbol{r}\, \psi_{\boldsymbol{k}_e}^*(\boldsymbol{r})\psi_{100}(r)e^{i\boldsymbol{q}\cdot\boldsymbol{r}}, \tag{5.89}$$

where $\psi_{100}$ is the ground state ($n = 1, \ell = 0, m = 0$) wave-function of the hydrogen, and the continuum outgoing state is $\psi_{\boldsymbol{k}_e} \propto \exp(-\boldsymbol{k}_e \cdot \boldsymbol{r})$ for $r \to \infty$. For heavier atoms (for instance for noble gas atoms commonly used in direct detection experiments) the two single electron wave functions in eq. (5.89) are replaced by the corresponding multi-electron wave-functions, and $e^{i\boldsymbol{q}\cdot\boldsymbol{r}} \to \sum_i^{N_e} e^{i\boldsymbol{q}\cdot\boldsymbol{r}_i}$, where the sum is over all the electrons in the atom. In the Hartree-Fock approximation the exact many-electron wave functions are approximated by Slater determinants of single-particle wave functions, of $N_e$ bound electrons in the initial state, and of $N_e - 1$ bound electrons and one continuum wave function in the final state [227].

Often we are interested in the spectrum of ionized electrons, i.e., in the differential rate $dR/dE_e$, where $E_e = \omega - E_B$ is the energy of the ionized electron. To obtain $dR/dE_e$ we use a quantity related to the dynamic structure function in eq. (5.88), but without evaluating the integral over the magnitude of the electron momentum, $k_e$. It is then convenient to define the *ionization form factor*

$$\big|f_{\mathrm{ion}}(k_e, q)\big|^2 = \sum_{\ell,m} \frac{2k_e^3}{(2\pi)^3}\big|f_{0\to k_e,\ell,m}(\boldsymbol{q})\big|^2, \tag{5.90}$$

where $f_{0\to k_e,\ell,m}(\boldsymbol{q})$ is given by a similar overlap integral as in eq. (5.89), but now with the outgoing wave function $\psi_{k_e,\ell,m}(\boldsymbol{r})$ that describes a spherical wave with a definite angular momentum quantum numbers $\ell, m$. That is, the outgoing wave function $\psi_{\boldsymbol{k}_e}$ is expanded in spherical waves, and thus the integral over $d^3\boldsymbol{k}_e$ in eq. (5.88) is replaced by a summation over $\ell, m$ and an integral over $dk_e$. This gives

$$\frac{dR_{\mathrm{ion}}^{\mathrm{SI}}}{dE_e} = \frac{1}{E_e}\frac{N_T}{M_T}\frac{\rho_\oplus}{M}\frac{1}{8\mu_e^2}\int dq\, q\, \bar{\sigma}(q)\big|f_{\mathrm{ion}}(k_e, q)\big|^2 \eta(v_{\min}), \tag{5.91}$$

where $\bar{\sigma}(q)$ is given in eq. (5.83), while the minimal DM velocity for obtaining an ionized state with an outgoing electron with energy $E_e$ is given by $v_{\min} = (E_e + E_B)/2 + q/2M$.

The scattering rates for ionization are suppressed by the relatively large exchange momenta compared to the atomic scales. Taking as an example the ionization from the ground state of the hydrogen atom in the large momentum exchange regime, $|\boldsymbol{q}| \gg |\boldsymbol{k}_e| \gg 1/a_0$, the ionization form factor is proportional to $|f_{\mathrm{ion}}| \propto (a_0 k_e)^3/(a_0 q)^8$, showing the parametric suppression from the wave function overlaps.

## 5.2.2   Semi-conductors

Semi-conductors (Si or Ge) are one of the most commonly used materials in direct detection experiments, both because it is relatively easy to produce pure detectors using semi-conductors, as well as because such detectors can potentially detect low mass dark matter, $M < 1$ MeV, when scattering on electrons (however, see below for challenges). The conventional semi-conductors have energy gaps between electronic bands that are $\mathcal{O}(\text{eV})$. This means that in DM scattering events the momentum exchange is always larger than the inverse distance between the atoms in the lattice, see eq. (5.87). The collective effects are thus expected to be less important for DM scattering on conventional semi-conductors.

Nevertheless, the valence electrons are delocalised in a crystal so that their wave-functions have support throughout the whole crystal, which needs to be taken into account when calculating DM scattering rates. Since we are mostly interested in the spin independent scattering on non-relativistic electrons we can use that the electromagnetic current $\langle e | \gamma^\mu | e \rangle \to (1_{\text{el}}, \mathbf{0})$ in the non-relativistic limit. The response of the material to the DM interaction $\mathcal{O}_1^{\text{el}} = 1_{\text{DM}} 1_{\text{el}}$ can therefore be related to the response of the material to the electromagnetic interactions, encoded in the dielectric function $\epsilon(\mathbf{q}, \omega)$. After some algebra one obtains [228, 229]

$$S(\mathbf{q}, \omega) = \frac{\mathbf{q}^2}{2\pi\alpha} \text{Im}\left( -\frac{1}{\epsilon(\mathbf{q}, \omega)} \right) \frac{1}{1 - \exp(-\omega/k_B T)}. \tag{5.92}$$

The energy loss function $\text{Im}\left[ -1/\epsilon(\mathbf{q}, \omega) \right] = \text{Im}\left[ \epsilon(\mathbf{q}, \omega) \right]/\left| \epsilon(\mathbf{q}, \omega) \right|^2$ describes the rate for the charged particle traversing the material to lose momentum $\mathbf{q}$ and energy $\omega$. Note that $\epsilon$ depends on both $\mathbf{q}$ and $\omega$, and thus generalizes the frequency dependent dielectric constant $\epsilon(\omega)$, which describes the response of a material to the propagating electromagnetic wave (the on-shell photons). In scattering of DM, on the other hand, $q \gg \omega$, see fig. 5.13 (left), so that $\epsilon(\mathbf{q}, \omega)$ describes the response of a material to the exchange of off-shell "potential photons". Sometimes the screening factor is approximated as $1/\left| \epsilon(\mathbf{q}, \omega) \right|^2 \simeq 1$, which is a reasonable approximations for $q \gtrsim$ keV, but numerically this still leads to a factor of a few difference in the predicted rates for scattering in Si and Ge.

Eq. (5.92) is very useful, since there are both experimental data on the energy loss function as well as established condensed matter techniques to evaluate the structure functions. The comparison between different theoretical approaches can then be used to estimate how large the uncertainties are. Fig. 5.13 (right) shows the prediction for dynamical structure function $S(\mathbf{q}, \omega)$ for Si, obtained using time dependent density functional theory. For energy deposits with $\omega \sim 10 - 20$ eV and low $|\mathbf{q}|$ there is a collective excitation — the plasmon. At higher momenta exchanges the dynamical response function peaks around the free-electron dispersion $\omega = \mathbf{q}^2/2m_e$ with a rather broad support. The region that can be accessed by DM scattering is below the diagonal dashed line, i.e., below the region where collective effects are important. The larger momenta and smaller energy deposits that characterise the DM scattering events therefore imply suppressed values of $S$.

## 5.2.3   Novel/low threshold materials

DM scattering either on electrons bound in atoms or on electrons inside the semi-conductors is not ideally kinetically matched, with the largest values of $S(\mathbf{q}, \omega)$ lying outside the accessible region of momenta and energy exchanges. Efforts devoted to finding more suitable target materials with smaller band gaps include the superconductors and Dirac materials, see fig. 5.13 (left).

Interestingly, superconductors were historically the first materials proposed for detection of dark matter recoils. The idea for the so-called superheated superconducting colloid detectors was that they would consist of superconducting grains of materials cooled just below the transition temperature [181]. DM scattering on a nucleus would then deposit enough energy to raise the temperature in the grain and trigger the transition from superconducting to normal phase, resulting in a signal that could be read out. This original detector concept was never used for DM detection, but did motivate searches for DM

nuclear recoils using other types of detectors based on semi-conductors and noble gases. The interest in superconductors as possible detector materials has been revived more recently with the increased interest in sub-GeV DM that could be searched for via scattering on electrons [230], with first bounds now appearing using superconducting nano-wires [222, 231, 232]. The electrons in a superconductor are bound into Cooper pairs, with typical binding energy in the meV regime. DM scattering on electrons would break Cooper pairs, resulting in a visible signal, potentially even for sub-MeV DM. The practical problem is that the deposited energy needs to be large enough not just to break the Cooper pair but to also pull the electron out from the Fermi sea, i.e., to avoid the Pauli blocking. Effectively, this means that the reach of superconducting targets to sub-MeV DM is limited by the screening effects encoded in the behaviour of the dielectric constant at small $q$ [228].

For DM with mass above MeV the conventional semi-conductors based on Ge and Si targets thus still remain the best. For semi-conductors there are no large screening effects in the low wavelength limit, $q \to 0$. This is a common feature for all materials that have a Fermi level lying between the valence and conducting bands. The sensitivity to sub-GeV DM is still cut-off, though, by the $\mathcal{O}(1\,\mathrm{eV})$ energy gap between the two bands in the semi-conductors, but could be improved with sensitivity to lower DM masses, if detectors were built from novel narrow-gapped semiconductor materials. An example of such a narrow-gap semiconductor is a gapped Dirac material [233], in which the conducting and valence bands have a linear dispersion relation, $E_{\pm} = \pm\sqrt{v_F^2 \boldsymbol{k}^2 + \Delta^2}$, the same as the dispersion relation for the relativistic fermions.[20] The appearance of the relativistic dispersion relation (in contrast to the non-relativistic one, $E \propto \boldsymbol{k}^2$) in a condensed matter system may seem surprising, since individual electrons are non-relativistic. However, the relativistic dispersion relation is not a sign of degrees of freedom propagating at the speed of light, but rather a result of being able to Taylor expand the energy levels linearly in $\boldsymbol{k}$ around special ("Dirac") point $\boldsymbol{k}_0$ at which the valence and conducting bands touch, if such a point exists. This gives the limit $\Delta \to 0$. A small band gap $\Delta \sim \mathcal{O}(\mathrm{meV})$ is obtained, if the lattice symmetry is broken, for instance, by applying strain on a Dirac material. The gap is important for practical purposes, since it suppresses thermal noise, a required property for constructing a viable DM detector [234]. Dirac materials also have a useful property: in general $v_F$ is different in different crystal directions, which can then be used to search for DM signal via daily modulation of the signal [235] (isotropic materials too can measure directionality, if for the scatterings on single quasiparticles the correlation between the direction of the DM wind and outgoing quasiparticle can be traced; an example was worked out for superconductors in [236]).

Some other ideas for DM direct detection are resonant DM absorption in molecules [237], via chemical bond breaking [238], and color centers in crystals [239]. The detection threshold can also be lowered by scattering on collective modes. The use of DM scattering on quasiparticles was discussed already in the '80 (for an early review see [240]). If DM couples to electron spin, this can for instance induce magnon excitations [241], while low energy DM-nucleon scattering can result in single or few phonon excitations, for instance in polar materials, see the discussion in section 5.3.

### 5.2.4   Dark matter absorption

If DM is a light boson, either a scalar of a vector, with a mass below keV, it can be absorbed in a material resulting in electron excitation [242]. The kinematics of DM absorption is very different from DM scattering, since now we have $\omega = M$ and $\boldsymbol{q} \approx 0$, i.e., $\omega \gg |\boldsymbol{q}|$. For $M \sim \mathcal{O}(10\ \mathrm{eV})$ the absorption of a bosonic DM in a semiconductor can thus excite a plasmon, resulting in an enhanced signal. For instance, if DM is a kinetically mixed dark photon, the absorption rate per unit target mass is given by

---

[20]We use the notation in which the energy is measured with respect to the midpoint between the conducting and valence bands, so that the gap is $2\Delta$. The states in the valence band carry negative energies, and correspond to anti-particles in the relativistic theory; $\Delta$ plays the role of the Dirac mass, while Fermi velocity $v_F$ plays the role of the speed of light.

the energy loss function at (effectively) zero momentum exchange,

$$R = \frac{\rho}{\rho_T}\varepsilon^2 \operatorname{Im}\left[\frac{-1}{\epsilon(0, M)}\right], \tag{5.93}$$

where the parameter $\varepsilon$ gives the strength of kinetic mixing between dark photon and the usual photon, see section 9.4.1 (here we use $M$ for DM mass as usual, and not $m_{A'}$). Optical measurements directly probe $\epsilon(0, M)$ and thus one can use data to predict the absorption rate.

### 5.2.5   Codes for DM electron scattering

There are several codes that predict DM–electron scattering rates. `QEdark` uses density functional theory to predict spin independent scattering on electrons in silicon, and is a companion code to the first direct detection limits on sub-GeV dark matter based on recasting the published results from DAMIC [243]. This was then extended to general DM EFT operators in `QEdark-EFT` (Catena et al. (2022) in [225], for fast parameter scanning there is also a machine learning based speed-up version `DEDD` available, Catena et al. (2024) in [225]). `DarkELF` [244] performs three different evaluations of the energy loss functions relevant for DM scattering on electrons for a number of targets, as well as the evaluations of the Migdal effect and dark matter phonon/scattering rates. The most sophisticated prediction uses time dependent density functional theory (the GPAW method), the most simplistic is based on approximating the material with a non-interacting Fermi liquid (Lindhard method), which is then also generalized to include dissipation and absorption (Mermin method). `EXCEED-DM` combines density functional theory calculations for states near the band gap and semi-analytic approximations for additional states farther away from the band gap to obtain predictions for DM–electron scattering rates in Si and Ge also for larger energy depositions [245]. `QCDark` calculates the dark matter-electron scattering rates in crystals using atom-centered gaussian basis, and also estimates systematic uncertainties [246]. A fast calculation of dark matter direct detection rates in the case of anisotropic materials is provided via the `VSDM` package [247].

## 5.3   Scattering on phonons

The formalism used to include material effects for sub-GeV DM scatterings on electrons also applies to scattering of sub-GeV DM on nucleons [223, 248]. For sub-GeV DM the momentum exchange between DM and the material becomes comparable or even smaller than the inverse atomic size, and nuclei no longer can be treated as free isolated particles. Depending on the DM mass there are several different regimes of detector response, illustrated in fig. 5.14. The red, green and orange shaded regions show the regions where the dynamical structure for spin-independent scattering on nucleons,

$$S(\boldsymbol{q}, \omega) = \frac{2\pi}{V}\sum_f \left|\langle f|\sum_I f_I e^{i\boldsymbol{q}\cdot\boldsymbol{r}_I}|i\rangle\right|^2 \delta(E_f - E_i - \omega), \tag{5.94}$$

has large support, and correspond to different physical processes as we discuss below. In eq. (5.94) the sum over $I$ runs over all the nuclei, with positions $\boldsymbol{r}_I$. In general, the target material is composed of several different atoms, for instance the Na and I for NaI crystals. The $f_I$ are the normalized interaction

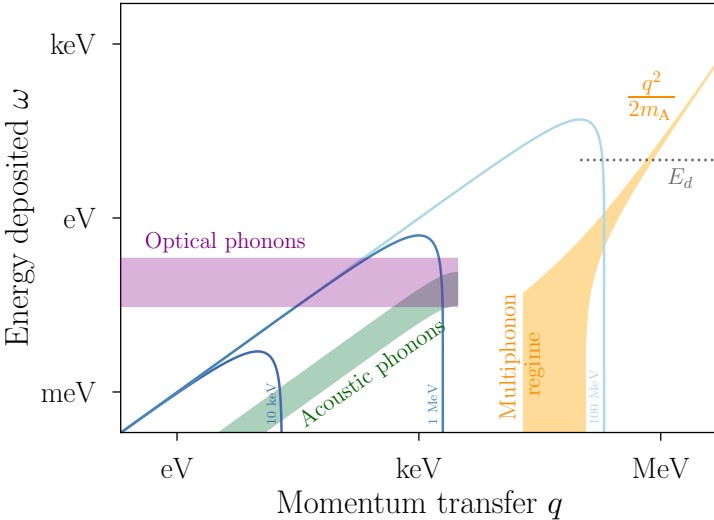

Figure 5.14: *Different kinematical regions relevant for DM scattering on nuclei. The free particle regime,* $\omega \simeq \boldsymbol{q}^2/2m_A$, *is approached for large momenta exchanges, and deposited energies well above the displacement energy for a nucleus,* $E_d$. *Adapted from [223] (©IOP Publishing, reproduced with permission).*

strengths for DM coupling to nucleon $I$,[21]

$$f_I = \left(\frac{2}{|c_1^p|^2 + |c_1^n|^2}\right)^{1/2} \left[Zc_1^p + (A - Z)c_1^n\right] F(q), \qquad (5.96)$$

where $c_1^N$ are the non-relativistic coefficients in the effective interaction Lagrangian, eq. (5.63), while $F(q)$ is the spin-independent nuclear form factor, cf. eq. (5.6). For isospin symmetric coupling, $c_1^p = c_1^n$, the normalized interaction strength is $f_I = A$ in the $q \to 0$ limit. Both $F(q)$ and $A, Z$ depend on the type of nucleus DM scatters on, so that in general $f_I$ changes from one type of nucleus to another, if DM couples differently to neutrons and protons, $c_1^p \neq c_1^n$. This means that for DM scattering on nucleons there is, for each of the target materials, not just one but a whole family of dynamical structure functions $S(\boldsymbol{q}, \omega)$, depending on which value the ratio $c_1^p/c_1^n$ takes.

For heavy DM, $M \gtrsim$ GeV, the energy exchange is bigger than the binding energy and thus the nuclei can be treated as free particles, as done in section 5.1. In this case the structure function is given by the free-nucleon limit up to nuclear corrections

$$S(\boldsymbol{q}, \omega) = 2\pi n_{\text{nuc}} A^2 \big|F(q)\big|^2 \delta\left(\omega - \frac{\boldsymbol{q}^2}{2m_A}\right), \qquad (5.97)$$

where for simplicity we assumed that the DM interaction is isospin symmetric, $c_1^p = c_1^n$. This is shown schematically in fig. 5.14 as the orange band. For energy deposits bigger than $E_d \sim \mathcal{O}(10\,\text{eV})$, the typical binding energy of atoms (nuclei) in molecules or condensed matter systems, the structure function

---

[21]Other normalizations are also used in the literature. For instance, if DM couples to both electrons and nuclei, the dynamical structure function is given by

$$S(\boldsymbol{q}, \omega) = \frac{2\pi}{V} \sum_f \Big|\langle f| \sum_k f_k e^{i\boldsymbol{q} \cdot \boldsymbol{r}_k} + \sum_I f_I e^{i\boldsymbol{q} \cdot \boldsymbol{r}_I} |i\rangle\Big|^2 \delta(E_f - E_i - \omega), \qquad (5.95)$$

where $f_k$ is the normalized spin-independent DM coupling to electrons and $f_I$ to nuclei. Conventionally, one often uses $f_e = 1$ and $f_I = \left[Z_I c_1^p/c_1^e + (A_I - Z_I)c_1^n/c_1^e\right]F(q)$. Taking as an example the dark photon, this gives $f_I = -Z_I$ in the long wavelength limit, $q \to 0$.

approaches the free particle dispersion relation centered at $\omega - \boldsymbol{q}^2/2m_A$. For DM with masses above $M \gtrsim \mathcal{O}(10\,\text{MeV})$ the binding energy of atomic systems becomes important, and needs to be taken into account. The typical momentum exchange $|\boldsymbol{q}| \sim Mv \gtrsim \mathcal{O}(10\,\text{keV})$, however, is large enough that the DM scattering can still be viewed as occurring mostly on a single bound nucleon. Viewed as a particle in a potential well, the nucleus behaves as a harmonic oscillator in the ground state. The scattering event excites the harmonic oscillator to a higher energy level, resulting in the spread of the dynamical structure function around the free particle dispersion relation, eventually leading to the multi-phonon regime for small enough momenta transfers.

For DM with masses below $\mathcal{O}(\text{MeV})$ the maximal momentum transfer is below $\sim 1$ keV. That is, the de Broglie wavelength of $|\boldsymbol{q}|$ is larger than the interatomic distance in a crystal lattice. For sub-MeV DM one therefore needs to take into account the couplings between different nuclei. DM scatters coherently on collective modes, phonons, which are the collective vibrational modes of the nuclei arranged in the crystal lattice. The so-called acoustic phonons have a linear dispersion relation $\omega_q = c_s|\boldsymbol{q}|$, where $c_s$ is the speed of sound. Acoustic phonons are massless Goldstone bosons associated with the spontaneous breaking of the translational symmetry. They are given by the oscillational modes that correspond in the $q \to 0$ limit to coherent shifts of all atoms in the same direction. Typically, crystals have more than one atom per unit cell so that other vibrational modes beside acoustic phonons also exist. The vibrational modes, where atoms within unit cell oscillate with the opposite phase, gives rise to optical phonons, i.e., phonons that are gapped.[22] Optical phonons are thus quasiparticles with nonzero mass $\omega_{\text{optical}} \simeq c_s/a \sim \mathcal{O}(10\text{s}\,\text{meV})$, where $a$ is the lattice spacing. The structure functions corresponding to single acoustic and single optical phonon excitations are denoted in fig. 5.14 as green and magenta bands, respectively.

For detection of DM scattering on phonons, other condensed matter systems, besides crystals, may possibly be used as well [249, 250]. In fact, the use of DM scattering off phonons as a possible detection technique for sub-GeV and sub-MeV DM was first considered for superfluid $^4$He (the proposal to use superfluid helium-4 as a DM detector material dates all the way back to 1988, though at that time not with sub-GeV DM in mind) [249]. For $M$ above few MeV the typical momentum exchange in DM scattering on superfluid helium is larger than the inverse of a typical distance between the atoms in the supefluid liquid, and the scattering can be viewed as scattering on individual atoms. For lighter masses, however, the collective effects are important, and DM scattering occurs on collective density fluctuations in the superfluid. The difference between semi-conductors and superfluid is that superfluid $^4$He is a strongly coupled system, and thus the predictions for the response function $S(\boldsymbol{q}, \omega)$ are much harder to come by. Beside the acoustic phonon contribution, the dispersion curve in superfluid He also contains two other excitations, the maxon and roton. To obtain $S(\boldsymbol{q}, \omega)$ one can try to use directly the data from neutron scattering, or theoretical tools used for strongly-correlated systems, including EFT methods (the latter however only capture the interactions with acoustic phonons). Since the speed of sound is lower in superfluid helium than in crystals, $c_2 \simeq 2.4 \cdot 10^2$ m/s, the DM scattering on a single acoustic phonon is not well kinematically matched. In most cases therefore the scattering of DM results in multiphonon excitations.

Another interesting possibility are polar materials, i.e., materials composed out of constituents that have nonzero dipole moments [250]. The gapped optical phonons in polar materials can be thought of as oscillating dipoles, which can thus have sizable couplings to certain types of dark matter, such as the kinetically mixed dark photons. The energy gap for optical phonons ranges from few meV to $\sim 100\,\text{meV}$, depending on the material. A useful property of polar materials is that the screening effects are also typically smaller than in superconductors and in Dirac materials. If the crystal is anisotropic, then DM

---

[22]Consider for instance the example where a unit crystal cell has two atoms. If these oscillate around their center of mass system, there is no net momentum flowing in and out of the unit cell, corresponding to a phonon with $\boldsymbol{q} = 0$, but with nonzero oscillation frequency (a.k.a. a mass gap). Such an oscillation, but modulated with a long wavelength off-set in the oscillations of the atoms in different cells, then gives an optical phonon with $\boldsymbol{q} \neq 0$.

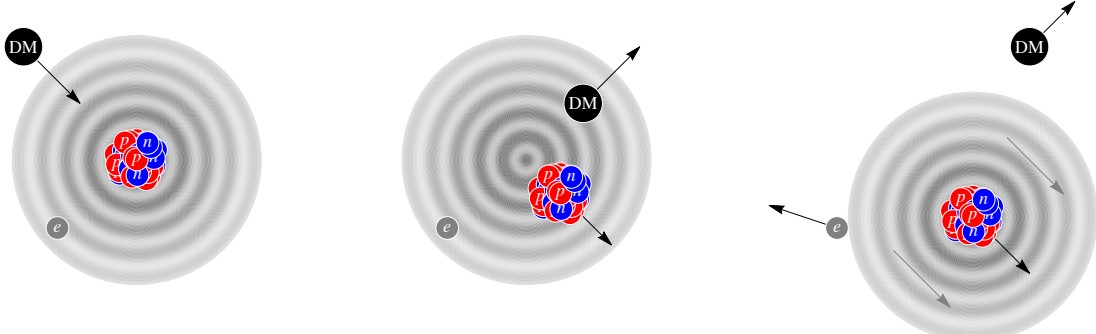

Figure 5.15: *Illustration of the atomic **Migdal effect**, where the scattering of DM on nucleus, and the corresponding nuclear recoil, cause the perturbation of the electronic wave functions, and results in an emission of an electron.*

scattering can also result in directional sensitivity. An interesting possibility discussed in the literature are metal halide perovskites, which are being developed in industry for high efficiency photo-voltaics, and could be repurposed for DM searches.

## 5.4   The Migdal effect

DM scattering on atoms can result in the process where DM first scatters on the nucleus, the nucleus recoils, this perturbation gets transferred to the electron cloud surrounding the nucleus, and one of the electrons gets kicked out [251]. This is the so called Migdal effect: it is a $2 \to 3$ process, with an incoming DM and atom scattering into the final state consisting of an outgoing DM particle, an ionized atom and a free electron. This is quite different from the elastic DM/nucleus (DM/atom) scatterings which were the topic of section 5.1. Elastic and inelastic DM/atom scatterings lead to different experimental signatures. While elastic DM/atom scatterings result in just the "nuclear recoils", the inelastic scatterings result in both the nuclear recoils and the free electrons (charge deposition) in the detector, more akin to the detector signatures due to DM scatterings on electrons. Since the detector thresholds for detecting electron excitations are typically lower, the Migdal effect can be exploited to extend the sensitivity of traditional DM detectors to DM/nucleus scattering for lighter DM masses.

The structure factor for the Migdal effect, assuming SI DM/nucleus scattering, is

$$\frac{dS_{\mathrm{Mig}}(\boldsymbol{q},\omega)}{d\omega_e} \simeq 2\pi n_{\mathrm{nuc}} A^2 \, \delta\!\left(\omega - \frac{\boldsymbol{q}^2}{2m_A} - \omega_e\right)\frac{dP}{d\omega_e}, \tag{5.98}$$

where $dP/d\omega_e$ is the probability distribution for the energy $\omega_e$ to be deposited in the electron excitation. In eq. (5.98) we assumed for simplicity isospin symmetric DM interactions, and set the nuclear form factor to its long wavelength limit, $F(q \to 0) = 1$, since experimentally the Migdal effect is most relevant for sub-GeV DM and thus small momenta exchanges. The expression (5.98) can be compared with eq. (5.97), i.e., the expression for the elastic DM–atom scattering. The factorized form of eq. (5.98) assumes that the momentum transfer is dominated by the nuclear recoil, with only a small fraction of the momentum carried by the free electron, which is easily satisfied for present experimental thresholds.

In the 'sudden recoil' approximation, i.e., assuming that the DM/nucleus collision occurs at time-scales much faster than the ones describing the dynamics of the electronic system, the initial electron

wave-function can be taken to be unperturbed. The probability distribution $dP/d\omega_e$ is then given by

$$\frac{dP}{d\omega_e} = \sum_f |\langle f| \exp\left(im_e \boldsymbol{v}_A \cdot \sum_k \boldsymbol{r}_k\right)|i\rangle|^2 \delta(E_f - E_i - \omega_e), \tag{5.99}$$

where $\boldsymbol{v}_A = \boldsymbol{q}/m_A$ is the velocity of the nucleus (and thus the ionized atom) after the recoil, and $\boldsymbol{r}_k$ are the positions of the electrons. The exponential phase factor is the net result of the nucleus receiving the kick from DM scattering, and is equivalent to boosting the initial electron wave-function $|i\rangle$ to the new rest frame of the nucleus. This simple expression was first obtained as a semi-classical approximation by Migdal in 1939, but can also be derived fully quantum mechanically [252] (for a quick but somewhat heuristic derivation highlighting the required change of variables see [223]). Note that the sum over electron positions is in the exponent of the phase factor, in contrast to the case of DM/electron scattering, eq. (5.84), where the number operator resulted in a sum over phase factors. The final states $\langle f|$ are products of the outgoing free electron wavefunctions and of the ionized atom, with the summation running over all the possibilities. For sub-GeV DM the recoiled nucleus is slow, $|\boldsymbol{v}_A| \ll 1$. Expanding the phase factors to linear order in $\boldsymbol{v}_A \cdot \boldsymbol{r}_k$ we see that the leading contribution to the Migdal effect is due to the dipole transition matrix elements. These can then be related to the measured photo-absorption cross sections.

For the atomic Migdal effect, the Migdal and electron ionization rates are schematically given by

$$\frac{dR_{\mathrm{Mig}}/d\boldsymbol{q}}{dR_{\mathrm{ion}}/d\boldsymbol{q}} > Z^2 \left(\frac{m_e}{m_A}\right)^2 (\boldsymbol{q} r_{\mathrm{atom}})^2, \tag{5.100}$$

where for simplicity we assumed equal couplings of mediators to protons and electrons. Here $r_{\mathrm{atom}} \sim a_0$ is the effective atomic radius. The rate for the Migdal effect is enhanced by $Z^2$, due to coherent scattering on nucleus (for isospin symmetric couplings of the mediator this would be replaced by $A^2$). On the other hand, the rate is suppressed by the small electron to nucleus mass ratio, $(m_e/m_A)^2$, which negates the $Z^2$ enhancement. Because of the $(\boldsymbol{q} r_{\mathrm{atom}})^2$ factor the Migdal spectrum is dominated by larger momentum transfers than electron ionisations. Taking all three factors into account we see that the rate for Migdal effect can be comparable or exceed the electron scattering rates for heavy atoms.

The Migdal effect can also occur in crystals, and describes the response of the electron distributions in a crystal due to perturbations from a single recoiling nucleus, resulting in collective electron excitations or ionisations [253]. In a crystal, the initial and final states are many body states, with valence electrons delocalised across the crystal. The electromagnetic interactions between the nucleus and a valence electron are screened by the bound inner-shell electrons and by all the other valence electrons in the material. The first effect can be modelled by introducing a $k$-dependent ion charge, $Z_{\mathrm{ion}}(k)$, while the second effect can be related to the energy loss function $\mathrm{Im}\left[-1/\epsilon(\boldsymbol{q}, \omega)\right]$ in a similar way as for the DM scattering on electrons, see section 5.2. The relative ionization vs Migdal rate in crystals is expected to be similar to eq. (5.100), valid for atoms.

Rather than triggering the emission of an electron, the DM/nucleus collision can instead result in an emission of a photon [254]. While this also provides a possibility to reduce the experimental energy thresholds, the corresponding scattering cross sections are smaller than for the Migdal effect, and thus less useful in practice.

The effect predicted by Migdal has not yet been observed. It has been searched for experimentally at energies relevant for DM detection in liquid Xenon using neutron scattering. The outcome was a puzzling null result where a positive signal was expected [255]. There are two possible resolutions: the somewhat less likely is that the theoretical calculations are overestimating the rate for the Migdal process, which would imply that all DM constraints relying on this process would need to be revised. The other option is that the electromagnetic interactions of the neutrons, used in the experiment, somehow complicate the

physics affecting the Migdal process. In such a case DM, due to its negligible electromagnetic interactions, would exhibit the expected Migdal effect.

## 5.5    Non-standard direct-detection signals

The above discussion of direct detection signatures assumed that DM scatters elastically, i.e., that DM does not change during the scattering. It also assumed that DM couples through a simple mediator to the SM. More and more complicated scenarios can be considered. Below, we discuss a sample of possible deviations from minimality with quite important phenomenological consequences.

### 5.5.1    Direct detection of inelastic DM

Instead of elastic scattering, $DM + SM \to DM + SM$, the DM particle could change during the scattering event to a different dark-sector state, $DM + SM \to DM' + SM$ [256]. If the new dark state is lighter, $M' < M$, the direct detection scattering becomes an *exothermic* reaction that can deposit significantly more energy in the detector than what is possible in elastic scattering, $\mu v^2/2$, where $v \sim 10^{-3}$ is the velocity of the DM particle, and $\mu = Mm/(M+m)$ is the reduced mass of the DM/nucleus system [257]. In the opposite case, $M' > M$, *endothermic* reactions arise, so that the direct detection scattering can even be kinematically completely forbidden, alleviating the direct detection constraints. Kinematic closure happens when $M' - M > \mu v^2/2 \sim \mathcal{O}(100\,\mathrm{keV})$. In such cases it is possible that the scattering is kinematically allowed on heavier nuclei and forbidden on light nuclei.[23] If DM scattering results in an inelastic nuclear transition, $DM + A^* \to DM' + A$, i.e., if DM up-scatters to the heavier state, while the initial nucleus $A^*$ transitions to a lower energy level $A$, one can search for mass splittings of up to $M' - M \sim \mathrm{MeV}$ (though, at present the experiments are sensitive only to very large scattering cross sections) [258].

Inelastic DM can be realized quite naturally. Let us consider, for example, a fermionic DM: a Dirac 4-component fermion $\Psi = (\psi_L, \bar{\psi}_R)$ is composed from two 2-component Weyl fermions $\psi_{L,R}$. In general, it can have a Dirac mass term, $M\bar{\Psi}\Psi = M\psi_L\psi_R + \mathrm{h.c.}$, as well as Majorana mass terms, $\delta_L\psi_L^2/2 + \delta_R\psi_R^2/2 + \mathrm{h.c.}$ In the limit of dominant Dirac mass terms, $M \gg \delta_{L,R}$, this is the so called "pseudo-Dirac" DM. The Majorana terms can naturally be small, since they break a possible global U(1) symmetry, the DM number. The Majorana terms, even if small, qualitatively change the mass eigenstates, which are then two quasi-degenerate Majorana fermions. In a 2-component notation they are given by

$$\chi_1 \simeq i\frac{\psi_L - \bar{\psi}_R}{\sqrt{2}}, \qquad m_1 \simeq M - (\delta_L + \delta_R)/2, \tag{5.101}$$

$$\chi_2 \simeq \frac{\psi_L + \bar{\psi}_R}{\sqrt{2}}, \qquad m_2 \simeq M + (\delta_L + \delta_R)/2. \tag{5.102}$$

In energetic enough scatterings, such that the energy transfer is well above $\delta_{L,R}$, the splitting can be neglected, and the two states behave as part of a single Dirac fermion. In the opposite, low-energy limit, the two states behave as two separate Majorana fermions.

The low energy limit also reveals a qualitative change in the structure of the interaction. Assuming a vector current interaction to quarks, $(\bar{\Psi}\gamma_\mu\Psi)(\bar{q}\gamma^\mu q)$, this corresponds to effectively elastic scattering at high energies (when mass eigenstates are degenerate to a good approximation), but to an inelastic interaction among mass eigenstates at low energies, transforming $\chi_1$ to $\chi_2$ in the scattering:

$$\bar{\Psi}\gamma_\mu\Psi \simeq i\big(\bar{\chi}_1\bar{\sigma}_\mu\chi_2 - \bar{\chi}_2\bar{\sigma}_\mu\chi_1\big). \tag{5.103}$$

---

[23]This property was the original motivation to introduce inelastic DM with $M' - M \sim 100\,\mathrm{keV}$ splitting between the states, which for a while lead to a viable explanation of the DAMA/LIBRA anomaly discussed in section 8.1.1.

Figure 5.16: **Bounds on sub-GeV** *Dark Matter. Boosted DM is discussed in section 5.5.4; BBN bounds in 6.12; White Dwarf bounds in 6.10.3. The astrophysical and cosmological bound is taken from Buen-Abad et al. (2022) [187], see section 5.1.2.*

A similar phenomenon happens, if a small mass term splits a complex scalar into two real scalars.

## 5.5.2 Baryonic DM and nucleon decay

Models of baryogenesis involving dark sectors (section 9.7) involve asymmetric DM carrying anti-baryonic global number, so that the net baryon number, combining visible and dark sectors, is zero. Antibaryonic DM can, in a very special case, lead to a spectacular signature in direct detection experiments — a DM-induced nucleon decay [259]. If the dark sector consists of two stable states, the anti-baryonic DM and a slightly lighter DM′ with no baryon number, then the DM can scatter inelastically in the target material, $DM + p \to DM' + \pi^+$ or $DM + n \to DM' + \pi^0$. Other mesonic final states with more pions or with kaons, are also possible; DM with baryonic number $1/3$ can induce $\overline{DM} + n \to DM + DM$. In all these cases, from observer's point of view, the proton $p$ or a neutron $n$ bound inside a nucleus would appear to decay, a signature usually associated with Grand Unified Theories. The bounds on such processes are comparable to the bounds on the proton decay itself, and correspond to effective nucleon lifetimes of $10^{29-32}$ years. Note that the mass difference between DM and DM′ cannot be too large since then decays $DM \to DM' + \bar{n}$ become kinematically open and DM is no longer stable.

## 5.5.3 Secluded DM

Secluded DM is a generic mechanism by which direct detection scattering cross sections can be suppressed, while DM is still a thermal relic, however, largely decoupled from the SM [260]. Secluded DM does not couple directly to DM, but rather interacts with some extra mediator particles, which then couple to the SM. In the early Universe the DM relic abundance is determined by annihilations to mediators, i.e., the mediators are lighter than the DM. The mediators then decay to SM particles, with couplings that can be very small — lifetimes as long as a fraction of a second are fine, since the mediators still decay before the BBN. Since the couplings of mediators to the SM can be very small, this gives highly suppressed direct detection signals, possibly well below any present or future sensitivities.

## 5.5.4   Boosted DM

DM gravitationally bound inside the galactic halo is non-relativistic, with velocity below the escape velocity $v_{\rm esc} \approx 10^{-3}$, see section 2.3. Different processes can accelerate a small fraction of DM particles to higher energies, generating a 'boosted DM' component. Some of the main possibilities are:

1. Multiple scatterings with fast moving nuclei or electrons inside the Sun, where DM gains kinetic energy and evaporates (the so called *solar reflection*). The energy transfer is large enough that sub-GeV DM can now have enough energy to be seen in direct detection experiments [261].

2. Collisions of DM particles with cosmic rays in the Galactic environment; this boosted component of the DM wind is sometimes called the *cosmic ray DM* [262].

3. A similar effect could arise around sources of cosmic rays, especially blazars such as the BL Lacertæ (the nearby active galactic nucleus that emits a jet pointing towards us). The resulting flux of boosted DM depends on the uncertain DM density around such sources [263].

4. The evaporation of Primordial Black Holes at present times [264]. This is analogous to the process that could have produced DM particles in the early Universe [174]. PBHs with masses from $10^{14}$ to $10^{16}$ g are a source of boosted MeV DM with energies of tens of MeV.

5. Decays or annihilations of heavier dark matter states; this possibility is, however, more speculative and model dependent [265].

Boosted DM is an especially important effect for sub-GeV DM. After boosting, the DM particle can have enough kinetic energy to give a detectable energy deposition in direct detection targets, above the experimental threshold. By taking into account this effect, the lower reach of DM direct detection experiments gets extended down to $M \approx 100$ keV DM masses [266], as illustrated in fig. 5.16. We also mention that, as a related application, Herrera and Murase (2023) [262] derived constraints on sub-GeV DM scatterings on protons and electrons via the requirement that cosmic rays emitted by AGN are not slowed down too much by the interactions with the surrounding DM.

## 5.5.5   Self-destructing DM

A long lived DM state might convert through scattering on nucleus or electron into a short-lived dark sector state [267]. This then decays to standard model particles within the detector, giving a visible deposited energy that can be comparable to DM mass — many orders of magnitude larger than the recoil energy deposited during elastic scatterings. This allows to search for self-destructing DM in large detectors with high energy thresholds, such as the neutrino detectors and not just with dark matter direct detection experiments.

## 5.5.6   Absorption of DM

If DM has interactions with the SM that involve only a single DM field, such as DM $\bar{q}q$ or DM $\bar{e}e$, then the DM particle can get absorbed in the target material when interacting with an electron or with a nucleon [268]. Such interactions also induce DM decays and therefore need to be highly suppressed so that DM is long-lived enough on cosmological time scales. This means that absorption of DM is a very rare event. The experimental signature, on the other hand, is quite different compared to DM scattering. In DM absorption the deposited energy is controlled by the DM mass, $M$, and is thus six orders of magnitude larger than the energy deposited in the scattering event (controlled by DM kinetic energy, $Mv^2/2 \approx 10^{-6}M$). Through DM absorption on electrons one can thus probe also very light bosonic DM, with masses in the $\mathcal{O}(10\,{\rm eV})$ range (see the more detailed discussion in section 5.2.3). Absorption is also

possible for fermionic DM, by emitting a light SM fermion in the process (similarly to interactions of SM neutrinos). Typically, the emitted particle is a neutrino, if absorption is through a neutral current interaction, or an electron, for a charged current interaction.

### 5.5.7 Multiple scatterings

The bounds on scattering cross sections of ultra-heavy DM, with mass above $M \gtrsim 10\,\text{TeV}$, are weak enough to allow events where DM scatters multiple times inside the detector [269] (the cross section has to be small enough that DM particles reach the underground detector, see also section 5.1.2). Such very heavy DM loses little energy in the scattering event and would travel through the detector in a straight line, leaving behind a track of small energy deposits. Such a signature is distinct enough that it can be distinguished from backgrounds. Dedicated analyses of data in conventional direct detection experiments are being performed to search for such events.

### 5.5.8 Scattering of composite DM

If DM is a composite object rather than a point-like particle, this can be accounted for through a DM form factor, $F_{\text{DM}}(q)$. In section 5.1.4 we took into account nuclear structure in the same way through the use of nuclear form factors. That is, the SI scattering cross section in eq. (5.6) now gets multiplied by $F_{\text{DM}}(q)^2$ (for SD scattering of composite DM one needs to introduce several form factors, the same as for nuclei in section 5.1.5). The DM form factor encodes the parametric suppression of scatterings in which momenta exchanges are large compared to the DM size, $r_{\text{DM}}$, $F_{\text{DM}}(q) \propto \exp(-q^2 r_{\text{DM}}^2/2)/(q r_{\text{DM}})^2$ for $q r_{\text{DM}} \gg 1$, see the expression for Helm nuclear form factor below eq. (5.12). Larger $r_{\text{DM}}$ implies loosely bound composite DM, in which case there is little momentum and energy transfer between DM and the target in the detector (imagine a ball of very soft foam scattering on a small hard target). The bounds from the usual direct detection experiments then become weaker. In an extreme situation, thousands or more DM particles can be bound in nuggets, held together by a Yukawa potential due to a light scalar mediator. Ref. [270] searched for such TeV mass nuggets, for mediators lighter than an eV, i.e., for Yukawa potentials with a range larger than about a $\mu$m. As a sensitive sensor they used an optically levitated sphere, and set bounds on DM-neutron scattering at the level of $\sigma_{\chi n} \lesssim 10^{-26}\,\text{cm}^2$ for nugget masses in the TeV range.

### 5.5.9 DM gravitational effects

DM with a Planck-scale mass that only interacts through the gravitational force can be searched for in the laboratory using an array of mechanical sensors, operated as a single detector. The passage of a DM particle with galactic velocity $v \sim 10^{-3}$ through the detector can potentially be measured, if DM is heavy enough, since DM exerts a gravitational force transferring the impulse

$$p_\perp = \int dt\, F_\perp = \tau \frac{G M m_{\text{detector}}}{b^2} \approx 10^{-20}\,\frac{\text{m}}{\text{s}}\, m_{\text{detector}} \left(\frac{\text{mm}}{b}\right)^2, \qquad (5.104)$$

where $\tau = 2b/v$ is the event duration and $b$ the impact parameter. This can be compared to the thermal fluctuation $\langle \delta p_\perp^2 \rangle \sim m_{\text{detector}} T \tau / \tau_{\text{detector}}$ for a single suspended detector with a mechanical damping time $\tau_{\text{detector}} \sim 10^8\,\text{s}$. The DM flux is large enough, if DM is light enough, $\Phi_{\text{DM}} = \rho_{\text{DM}} v/M \approx 1/(\text{yr}\,\text{m}^2)(M_{\text{Pl}}/M)$. This means that DM with a Planck-scale mass requires detectors with a cross sectional size of about a square meter. To obtain a measurably large signal will require about $10^6$ gram-size accelerometers. A development of such a sensor was proposed by the WINDCHIME collaboration [271]. The same type of detectors could search for DM with speculative extra long-range forces.

| Experiment | Location | Data taking | Readout | Target | Home | Ref. |
|---|---|---|---|---|---|---|
| PNL/USC | Homestake, SD | 1986 → 1991 | ioniz. (∼77 K) | 0.7 → 2 kg Ge | paper | [272] |
| UCSB/LBL/UCB | Oroville dam, CA | 1986 → 1990 / 1989 | ioniz. (∼77 K) / ioniz. (77 K) | 0.9 kg Ge / 17 g Si | paper | [273] |
| GOTTHARD GE | Gotthard, Switzer. | 1986 → 1990 | ioniz. (∼77 K) | 0.7 kg Ge | web | [274] |
| COSME | Canfranc, Spain | 1990 → 1999 | ioniz. (∼77 K) | 234 g Ge | paper | [275] |
| HEIDEL.-MOSCOW | Gran Sasso, Italy | 1990 → 1998 | ioniz. (∼77 K) | 2.76 kg Ge | web | [276] |
| EDELWEISS-0 | Modane, France | 1991 → 1994 | ther. phon. (50 mK) | 24 g Al$_2$O$_3$ | paper | [277] |
| TWIN | Homestake, SD | 1992 | ioniz. (∼77 K) | ∼2.1 kg Ge | paper | [278] |
| BRS | Gran Sasso/Modane | 1992 | scint. (∼300 K) | 0.7 kg NaI(Tl) | paper | [279] |
| NAI32 | Canfranc, Spain | 1992 → 1995 | scint. (∼300 K) | 32 kg NaI(Tl) | paper | [280] |
| ELEGANT V | Kamioka, Japan / Oto-Cosmo, Japan | 1992 → 1998 / 1999 → 2000 | scint. (∼300 K) | 36.5 → 662 kg NaI(Tl) / 730 kg NaI(Tl) | paper | [281] |
| BPRS | Gran Sasso, Italy / Gran Sasso/Modane | 1993 → 1994 / 1993 | scint. (∼300 K) / scint. (∼300 K) | 7 kg NaI(Tl) / 0.37 kg CaF$_2$(Eu) | paper | [282] |
| DEMOS | Sierra Grande, Argen. | 1994 → 1997 | ioniz. (∼77 K) | ∼ 1 kg Ge | paper | [283] |
| UKDM-NAI | Boulby, UK | 1994 → 1997 | scint. (∼300 K) | 1.3 → 6.2 kg NaI(Tl) | paper | [284] |
| MIBETA | Gran Sasso, Italy | 1995 → 2000 | ther. phon. (10 mK) | 0.34 kg TeO$_2$ | web | [285] |
| DAMA/LXE | Gran Sasso, Italy | 1995 → 2001 | scint. (170 K) | 6.5 kg LXe | web | [286] |
| HDMS | Gran Sasso, Italy | 1996 → 2003 | ioniz. (∼77 K) | 0.2 kg Ge | web | [287] |
| DAMA | Gran Sasso, Italy | 1996 → 2002 | scint. (∼300 K) | ∼ 100 kg NaI(Tl) | web | [288] |
| CDMS-I | Stanford, CA | 1996 → 1999 / 1996 → 1999 | ther. phon.+ioniz. (20 mK) / ath. phon.+ioniz. (20 mK) | 0.5 → 1 kg Ge / 0.1 → 0.2 kg Si | web | [289] |
| EDELWEISS-I | Modane, France | 1997 → 2004 | ther. phon.+ioniz. (∼ 20 mK) | 70 g → 1 kg Ge | web | [290] |
| SACLAY-NAI | Modane, France | 1997 | scint. (∼300 K) | 10 kg NaI(Tl) | paper | [291] |
| ELEGANT-VI | Oto Cosmo, Japan | 1998 → 2007 | scint. (∼300 K) | 10 kg CaF$_2$(Eu) | paper | [292] |
| IGEX-DM | Canfranc, Spain | 1999 → 2002 | ioniz. (∼ 77 K) | 2.0 kg Ge | paper | [293] |
| ROSEBUD | Canfranc, Spain | 1999 → 2002 / 1999 → 2002 / 1999 → 2002 | ther. phon. (20 mK) / ther. phon.+scint. (20 mK) / ther. phon. (20 mK) | 50 g Al$_2$O$_3$ / 54 g CaWO$_4$ / 67 g Ge | paper | [294] |
| SIMPLE | LSBB, France | 1999 → 2010 | superheat. dropl. (∼290 K) | 15 → 215 g C$_2$ClF$_5$ | paper | [295] |
| CRESST-I | Gran Sasso, Italy | 2000 | ther. phon. (15 mK) | 262 g Al$_2$O$_3$ | paper | [296] |
| NAIAD | Boulby, UK | 2000 → 2003 | scint. (∼ 280 K) | 46 → 55 kg NaI(Tl) | paper | [297] |
| TOKYO-DM | Kamioka, Japan | 2001 → 2002 / 2002 → 2003 / 2005 | ther. phon. (10 mK) / ther. phon. (4 K) / scint. (∼ 300 K) | 168 g LiF / 176 g NaF / 310 g CaF$_2$(Eu) | paper | [298] |
| ZEPLIN-I | Boulby, UK | 2001 → 2003 | scint. (∼ 170 K) | 3.2 kg Xe | web | [299] |
| CDMS II | Stanford, CA / Soudan, MN | 2001 → 2002 / 2003 → 2008 | ath. phon.+ioniz. (20 mK) / ath. phon.+ioniz. (50 mK) | 1 kg Ge / 0.2 kg Si / 1 → 4.6 kg Ge / 0.2 → 1.2 kg Si | web | [300] |
| ORPHEUS | Bern, Switzerland | 2002 → 2003 | superheat. grains (∼100 mK) | 0.45 kg Sn | paper | [301] |
| KIMS | Yangyang, S. Korea | 2004 → 2012 | scint. (∼ 300 K) | 6.6 → 103.4 kg CsI(Tl) | web | [302] |
| PICASSO | SNOLAB, Canada | 2004 → 2014 | superheat. dropl. (∼ 300 K) | 0.02 → 3.0 kg C$_4$F$_{10}$ | web | [303] |
| COUPP | Fermilab, IL / SNOLAB, Canada | 2005 → 2009 / 2010 → 2011 | bubble chamber (∼ 300 K) / bubble chamber (303 K) | 1.5 → 3.5 kg CF$_3$I / 4 kg CF$_3$I | web | [304] |
| ZEPLIN-II | Boulby, UK | 2006 | scint.+ioniz. (∼ 170 K) | 7.2 kg Xe | paper | [305] |
| XENON10 | Gran Sasso, Italy | 2006 → 2007 | scint.+ioniz. (179 K) | 5.4 kg Xe | web | [306] |
| NEWAGE | Kyoto, Japan / Kamioka, Japan | 2006 / 2008 → 2018 | ioniz. (∼ 300 K) / ioniz. (∼ 300 K) | ∼ 10 g CF$_4$ / ∼ 10 g CF$_4$ | web | [307] |
| TEXONO | Kuo-Sheng, Taiwan | 2007 → 2013 | ioniz. (∼ 77 K) | 20 → 840 g Ge | paper | [308] |

Table 5.1: ***Historic Direct Detection experiments*** *that placed bounds on DM nuclear scattering, ordered by the year of the first physics run (3rd column). The 2nd column shows the location, the 4th column the technology used, the 5th column the fiducial target mass, and the last two columns the references. The abbreviations for the readout channels are: ath. phon. → athermal phonons, ther. phon. → thermal phonons, scint. → scintillation light, ioniz. → ionization, superheat. dropl. → superheated droplets, superheat. grains → superheated grains. See table 5.2 for the modern experiments.*

| Experiment | Location | Data Taking | Readout | Target | Home | Ref. |
|---|---|---|---|---|---|---|
| DAMA/LIBRA | Gran Sasso, Italy | 2003 → 2024 | scint. (273 K) | 250 kg NaI(Tl) | web | [309] |
| EDELWEISS II | Modane, France | 2008 → 2010 | ther. phon.+ioniz. (18 mK) | 4 kg Ge | web | [310] |
| ZEPLIN-III | Boulby, UK | 2008 → 2011 | scint.+ioniz. ($\sim$ 170 K) | 6.5 → 5.1 kg Xe | web | [311] |
| CoGeNT | Chicago, IL | 2008 | ioniz. ($\sim$ 90 K) | 475 g Ge | paper | [312] |
| | Soudan, MN | 2009 → 2013 | ioniz. ($\sim$ 90 K) | 330 g Ge | | |
| CRESST-II | Gran Sasso, Italy | 2009 → 2015 | ther. phon.+scint. (10 mK) | 2.7 → 5 kg CaWO$_4$ | web | [313] |
| XENON100 | Gran Sasso, Italy | 2009 → 2014 | scint.+ioniz. (182 K) | 48 → 34 kg Xe | web | [314] |
| DRIFT-II | Boulby, UK | 2009 → 2016 | ioniz. ($\sim$ 300 K) | 100 g CS$_2$+38 g CF$_4$ | web | [315] |
| DMTPC | MIT, MA | 2010 | ioniz. ($\sim$ 300 K) | 3.3 g CF$_4$ | paper | [316] |
| DAMIC | Fermilab, IL | 2010 → 2011 | ioniz. ($\sim$ 140 K) | 0.5 g Si | web | [317] |
| | SNOLAB, Canada | 2014 → 2018 | ioniz. ($\sim$ 120 K) | 7 → 40 g Si | | |
| DM-Ice17 | South Pole | 2011 → 2015 | scint. ($\sim$ 260 K) | 17 kg NaI(Tl) | web | [318] |
| CDEX-0 | Jinping, China | 2012 → 2013 | ioniz. ($\sim$ 90 K) | 20 g Ge | web | [319] |
| CDEX-1 | Jinping, China | 2012 → 2018 | ioniz. ($\sim$ 90 K) | 0.9 kg Ge | web | [320] |
| XMASS-I | Kamioka, Japan | 2012 → 2017 | scint. (173 K) | 832 kg Xe | web | [321] |
| CDMSlite | Soudan, MN | 2012 → 2014 | ath. phon.+ioniz. ($\sim$ 50 mK) | 0.6 kg Ge | web | [322] |
| SuperCDMS Soudan | Soudan, MN | 2012 → 2014 | ath. phon.+ioniz. ($\sim$ 50 mK) | 9 kg Ge | web | [323] |
| KIMS-NaI | Yangyang, S. Korea | 2013 → 2015 | scint. ($\sim$ 300 K) | 17 kg NaI | web | [324] |
| DarkSide-50 | Gran Sasso, Italy | 2013 → 2018 | scint.+ioniz. ($\sim$ 80 K) | 46 kg Ar | web | [325] |
| LUX | Sanford, SD | 2013 → 2016 | scint.+ioniz. ($\sim$ 180 K) | 145 → 98 kg Xe | web | [326] |
| PICO-2L | SNOLAB, Canada | 2013 → 2015 | bubble chamber (289 K) | 2.9 kg C$_3$F$_8$ | web | [327] |
| PICO-60 | SNOLAB, Canada | 2013 → 2014 | bubble chamber ($\sim$ 310 K) | 37 kg CF$_3$I | web | [328] |
| | | 2016 → 2017 | bubble chamber ($\sim$ 290 K) | 52 kg C$_3$F$_8$ | | |
| EDELWEISS-III | Modane, France | 2014 → 2015 | ther. phon.+ioniz.(18 mK) | $\sim$ 5 kg Ge | web | [329] |
| PandaX-I | Jinping, China | 2014 | scint.+ioniz. (179.5 K) | 54 kg Xe | web | [330] |
| PandaX-II | JinPing, China | 2015 → 2018 | ioniz.+scint. ($\sim$ 165 K) | $\sim$ 330 kg Xe | web | [331] |
| NEWS-G : SEDINE | Modane, France | 2015 | ioniz. ($\sim$ 300 K) | 280 g Ne+3 g CH$_4$ | web | [332] |
| XENON1T | Gran Sasso, Italy | 2016 → 2018 | ioniz.+scint. (177 K) | 1.0 → 1.3 t Xe | web | [185] |
| CRESST-III | GranSasso, Italy | 2016 → 2018 | ther. phon.+scint. (15 mK) | 24 g CaWO$_4$ | web | [333] |
| | | 2021 | ther. phon.+scint. (5 mK) | 20 g LiAlO$_2$ | | |
| DEAP-3600 | SNOLAB, Canada | 2016 → | scint. ($\sim$ 85 K) | 3.3 t Ar | web | [334] |
| COSINE-100 | Yangyang, South Korea | 2016 → 2023 | scint. ($\sim$ 300 K) | 61.3 kg NaI(Tl) | web | [335] |
| ANAIS-112 | Canfranc, Spain | 2017 → | scint. ($\sim$ 300 K) | 112.5 kg NaI(Tl) | web | [336] |
| CRESST proto. | Munich, Germany | 2017 | ath. phon.+scint. (10 mK) | 0.5 g Al$_2$O$_3$ | web | [337] |
| | | 2019 | ther. phon.+scint. (22 mK) | 2.7 g Li$_2$MoO$_4$ | | |
| CDEX-10 | Jinping, China | 2017 → 2018 | ioniz. ($\sim$ 90 K) | 0.9 kg Ge | web | [338] |
| EDELWEISS-SubGeV | Lyon, France | 2018 | ther. phon. (10 mK) | 33.4 g Ge | web | [339] |
| | Modane, France | 2018 → 2020 | ther. phon. (44 mK) | 200 g Ge | | |
| NEWS-G : SNOGLOBE | Modane, France | 2019 | ioniz. ($\sim$ 300 K) | $\sim$ 100 g CH$_4$ | web | [340] |
| | SNOLAB, Canada | 2022 → | ioniz. ($\sim$ 300 K) | $\sim$ 2 kg CH$_4$ | | |
| SuperCDMS-CPD | SLAC, CA | 2020 | ath. phon. (8 mK) | 10.6 g Si | web | [341] |
| SENSEI | Fermilab, IL | 2020 | ioniz. (130 K) | 1.9 g Si | web | [342] |
| | SNOLAB, Canada | 2022 → | ioniz. (130 K) | $\sim$ 100 g Si | | |
| LZ | Sanford, SD | 2021 → | ioniz.+scint. (174 K) | 5.5 t Xe | web | [184] |
| PandaX-4T | Jinping, China | 2021 → | ioniz.+scint. ($\sim$ 170 K) | 2.7 t Xe | web | [186] |
| XENONnT | Gran Sasso, Italy | 2021 → | ioniz.+scint. (177 K) | 4.4 t Xe | web | [343] |
| TESSERACT-Si | Berkeley, CA | 2024 | ath. phon. (48 mK) | 0.22 g Si | web | [344] |
| DAMIC-M | Modane, France | 2024 → | ioniz. ($\sim$ 120 K) | 26.4 g Si | web | [396] |

Table 5.2: ***Recent and presently running direct detection experiments*** *that placed bounds on* **DM scattering on nuclei.** *Experiments are ordered by the year of the first physics run (3rd column), with the 2nd column showing the location, the 4th column the signal channel used, and the 5th column the fiducial target mass. The last two columns give references for each experiment. The abbreviations used are the same as in table 5.1, which lists the historic experiments. For planned experiments see table 5.4.*

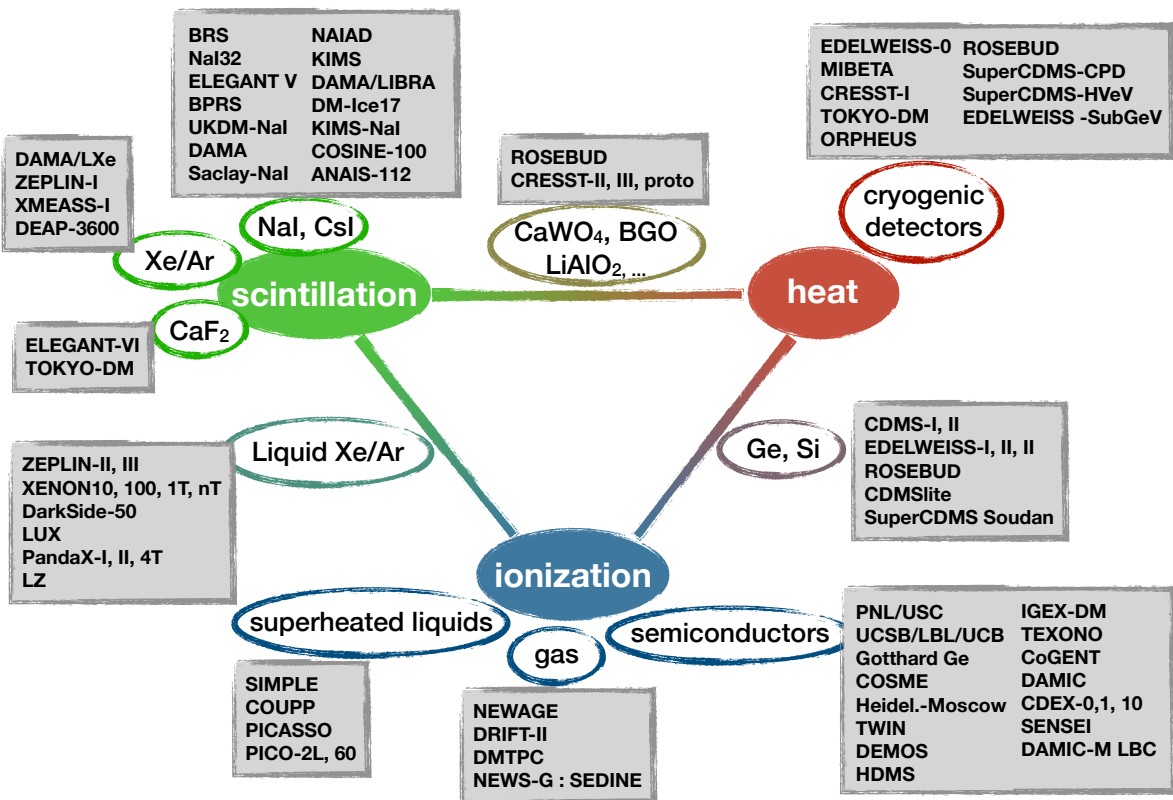

Figure 5.17: *An illustration of different* **techniques for DM direct detection**, *relying on three different ways of measuring the recoil energy deposited in the scattering on target: scintillation, heat and ionization.*

## 5.6   Experimental techniques

The main idea behind direct detection is to observe the energy deposited by DM scattering on matter. The two main scattering processes used in the present day experiments are: DM scattering off electrons (section 5.2) or off nuclei (section 5.1). For DM scattering on electrons the deposited energy is typically $E_R \sim$ eV, while for scattering on nuclei it is in the range of $E_R \sim$ few keV to 100s keV. The experimental techniques based on nuclear scattering have a longer history, so we focus on these first. The experimental techniques for observing scattering on electrons are then reviewed in section 5.6.7.

Experiments so far found that the rate of DM scattering elastically on nuclei is below few events per ton per year. This is much smaller than the typical rates from unshielded backgrounds, most commonly the $\gamma$ and $X$-rays, which lead to energy depositions in the form of electron recoils. The rates for these can be as high as kHz per kg of the detector mass. A significant amount of experimental attention is devoted to reducing the environmental backgrounds. This can be achieved by shielding against cosmogenic or environmental radioactivity, by purifying the active substance and by utilising the signal shape to discriminate between electron and nuclear recoil events. It is especially important to reduce the neutron backgrounds, because neutron scattering can mimic the DM induced nuclear recoils. For further discrimination one can use the fact that the DM scattering signal is expected to exhibit an annual modulation due to the Earth moving around the Sun, and the diurnal modulation due to the rotation of the Earth (see also section 5.1.7).

The energy deposited in DM scattering is observed either as *heat, scintillation*, or *ionization*. That is, the struck nucleus travels through the material and looses energy either by inducing motion of other

atoms in the target (heat — in crystals these are the thermal phonons), while a smaller amount of energy loss goes toward exciting or ionizing target atoms. The de-excitation of atoms leads to scintillation light. Either of the three channels can be picked up by appropriate sensors. While historic experiments relied on just one type of signal, many of the more recent experiments read two out of three, and none all three, see fig. 5.17 and tables 5.1, 5.2, 5.4.

Furthermore, the response of the detector is different for scattering on electrons and nuclei. The struck electrons lose energy more slowly while traversing the material, and induce almost exclusively a combination of ionization and scintillation. For detectors sensitive to ionization and/or scintillation but not heat, an electron recoil results in a much higher visible signal than a nuclear recoil event does (in the latter most of the energy goes into heat). For such detectors it is conventional to present results in terms of

$$E_{\mathrm{ee}}(\mathrm{keV_{ee}}) = Q_F \times E_R(\mathrm{keV_{nr}}), \tag{5.105}$$

where $E_{\mathrm{ee}}$, quoted in $\mathrm{keV_{ee}}$ (keV electron-equivalent), is the would-be energy of an electron recoil that would have resulted in the same detector response as the nuclear recoil of energy $E_R$ (quoted in $\mathrm{keV_{nr}}$). The ratio between the two defines the quenching factor, $Q_F$, which in general depends on $E_R$. The reason for this bizarre choice of units is that it is straightforward to obtain the response of a detector to electron recoils by using one of several $\gamma$ ray sources of well known energy. This gives the energy calibration of the detector in $\mathrm{keV_{ee}}$. Obtaining the detector response to nuclear recoils is more involved. One either needs to perform a dedicated measurement using a mono-energetic neutron source or obtain $Q_F$ from theory. For detectors that are sensitive to the deposited heat, the quenching factor in eq. (5.105) is irrelevant, and the results can be quoted directly using $E_R$.

The technologies used in DM direct detection can be grouped in terms of cryogenic detectors, scintillating crystals, noble liquid detectors, superheated liquids, and directional detectors, see also fig. 5.17. Next, we will discuss in more detail each of these technologies (for other reviews see the Direct Detection part of the reference list in [1]).

## 5.6.1 Cryogenic detectors

Cryogenic detectors are kept at temperatures well below the room temperature. They can accurately measure the energy transferred to the nucleus after the DM scattering event. The relative importance of the three signals — phonons, ionization and scintillation — depends on the detector material, the detector design, as well as on the read-out. For instance, for a very large detector both ionization and scintillation eventually convert to heat, i.e., phonons. Furthermore, the distribution of the signal among phonons, ionization and scintillation is different for DM (or neutron) scattering and for electron scattering. Measuring a signal in two out of three channels is thus an efficient way of reducing the backgrounds.

### Ionization detectors

The pioneering DM detectors, and some of the more recent cryogenic detectors such as TEXONO and CoGeNT, were operated as single parameter detectors, utilizing only the ionization channel, see table 5.1. These were the first detectors to search for DM through direct detection, starting in the late 1980s, as repurposed detectors, originally used in the searches for neutrinoless double $\beta$ decay. These ionization detectors are semiconductor detectors made out of germanium, operated at the liquid nitrogen temperature of $77\,\mathrm{K}$. The energy from nuclear recoil in the DM scattering event partially goes to excite electrons from the valence band to the conduction band. For instance, the quenching factor in Ge is about $Q_F \sim 25\%$. This means that an ionization signal of $1\,\mathrm{keV_{ee}}$ in the observed energy comes from a nuclear recoil energy of $4\,\mathrm{keV_{nr}}$. The excitation energy required to create a single electron–hole pair is low, $\mathcal{O}(1\,\mathrm{eV})$, so that a typical DM scattering event results in many charge carriers, about $\mathcal{O}(300/\mathrm{keV})$. Because of this, the ionization detectors have good energy resolution and low energy thresholds. They

apply a high voltage $\sim$ keV, to collect all the deposited charge, which, on the other hand, makes difficult the measurements of the heat deposition in terms of phonons.

### Ionization phonon bolometers

The CDMS and EDELWEISS collaborations developed Ge and Si detectors that simultaneously measure the ionization and phonon signals. Since the detectors are operated at mK temperatures a discrimination between nuclear and electron recoils is possible even for few keV recoil energies. An important systematic is the incomplete charge collection from the scatterings that occur close to the surface. The collaborations were able to reduce this background with special interleaved electrode readouts. The new SuperCDMS detectors are expected to reach low thresholds, of $10\,\mathrm{eV_{ee}}$ for phonons.

### Scintillation phonon bolometers

The CRESST collaboration developed detectors on $CaWO_4$ crystals that can measure both the scintillating and phonon signals. The light yield, the ratio of the scintillation and phonon signals, is used to discriminate different scattering types — the neutron or DM scattering from the electron or $\gamma$ backgrounds. DM can scatter on any of the three nuclei, calcium ($Z = 20, A \approx 80$), tungsten ($Z = 74, A \approx 180$), or oxygen ($Z = 8, A = 16$), so that one detector combines light and heavy nuclear targets. The light yield for DM scattering differs slightly for each of the three cases, which needs to be taken into account in the analysis. The main goal of the CRESST collaboration in recent years was to lower the energy thresholds to below $100\,\mathrm{eV_{nr}}$ in order to increase sensitivity to sub-GeV DM.

## 5.6.2    Scintillating crystals

The scintillating inorganic crystal used in DM detection is most commonly NaI(Tl), and, less commonly, CsI(Tl). Both of these types of detectors are usually operated at room temperature. The DM detection mechanism relies on the fact that part of the nuclear recoil energy from the DM scattering event is converted into scintillation light. When the recoiled nucleus traverses the scintillating material, it excites the atoms, which then de-excite by emitting light. The NaI and CsI crystals are doped with thalium. This creates defects in the crystal lattice, increasing the light output of the scintillator. It also shifts the emitted light to longer wavelength, at which the crystals are more transparent, while at the same time the light can also be detected with higher efficiency using photosensors.

The detectors based on NaI(Tl) and CsI(Tl) scintillators have several advantages. The crystals have high mass density, so that it is possible to build targets with large mass. They have high light output, which translates into good energy resolution, and have lower energy thresholds than other scintillators. The detectors are also relatively simple, making it easier to achieve stable operating conditions over periods of several years. The disadvantage is that the energy deposition is measured only in one channel, scintillation, so it is harder to reject backgrounds. This is not possible on event by event basis, but only statistically, using the fact that the DM signal should exhibit an annual modulation, which for the spin-independent interactions would peak at or around June 2nd.

Such a signal is claimed with $13\sigma$ significance in the combined DAMA and DAMA/LIBRA datasets obtained using the ultra low radioactive NaI(Tl) crystals operated at the LGNS laboratory over a period of 14 annual cycles [288, 309]. A comparison with the other experiments that do not see a DM signal excludes the DAMA measurement to be due to DM, see section 8.1.1. In particular, a model-independent check of DAMA signal is possible using the same inorganic scintillator, NaI(Tl). Such detectors are DM-Ice17 operating at the South Pole since 2011 [318], the COSINE experiment at Yangyang, South Korea and the experiment SABRE in construction, with two detectors, in the south and north hemispheres [345]. If DM is assumed to scatter on iodine, a model independent check is provided also by CsI(Tl) scintillators, used in the KIMS experiment at Yangyang, S. Korea [302].

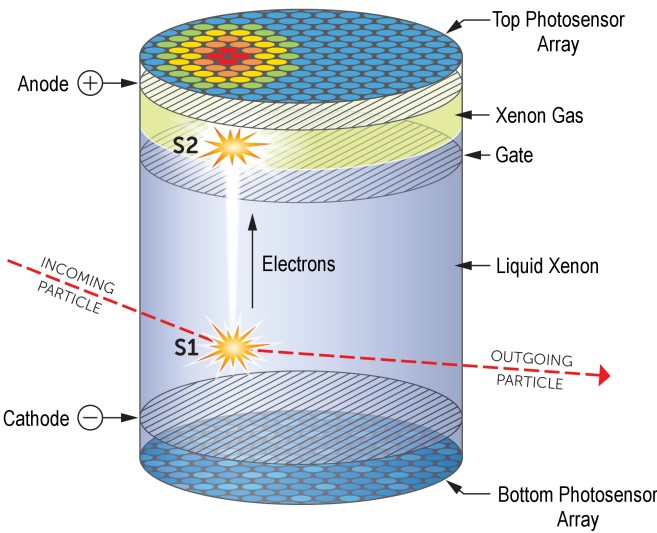

Figure 5.18: *The schematics showing the principles of a* **noble liquid/noble gas dual phase TPC detector***, giving prompt S1 scintillation and delayed S2 ionization signals. Figure from J. Aalbers et al. (2022) in [347].*

### 5.6.3 Noble liquid detectors

For a large range of DM masses the most stringent constraints on DM scattering at present come from the detectors using noble liquids, see fig.s 5.5 and 5.6. These type of detectors have several advantages, including a reasonably straightforward scalability to bigger target masses. An important asset of detectors based on liquid xenon (LXe) and liquid argon (LAr) is that they can measure the signal from DM scattering in two channels, ionization and scintillation. The scintillation signal originates from excited atoms that form dimers, $Xe_2^*$ and $Ar_2^*$. These de-excite to a dissociate ground state, 2Xe and 2Ar, emitting in the process photons with wavelengths of 178 nm and 128 nm, respectively. Scintillation efficiency is an important input describing how detectors respond to DM scattering, and is measured in dedicated experimental set-ups. The relative scintillation efficiency, $L_{\text{eff}}$, gives the scintillation light yield of nuclear recoils compared to electron recoils of the same energy. It is significantly lower than 1, signifying that electron recoils are much more efficient in producing the scintillation signal. Furthermore, $L_{\text{eff}}$ decreases for smaller recoil energies. It equals, for instance, $L_{\text{eff}} \approx 0.1$ in Xe for $E_{\text{nr}} \approx 10$ keV, see, e.g. [326, 346]. For very low recoil energies, $E_{\text{nr}} \approx 1$keV, $L_{\text{eff}}$ is still rather uncertain, and is an important systematic uncertainty when interpreting the scattering cross section bounds for DM masses below $M \lesssim 10$ GeV. In contrast, the ionization charge yield of nuclear recoils, in $e^-/\text{keV}_{\text{nr}}$, increases for low-energy recoils. This helps in detecting recoils of GeV or sub-GeV DM.

A typical design of the LXe (or LAr) detectors is a two-phase time projection chamber (TPC). The bulk of the detector is LXe, with gaseous Xe layer on top, see fig. 5.18. DM scattering on LXe in the active volume of the TPC creates both scintillation light and the ionization electrons. The scintillation light is collected by the photo-multiplier tubes (PMTs) at the perimeter of the detector, giving the prompt "$S1$" signal. Under the influence of external electric field, the ionization electrons drift in pure LXe toward the gaseous Xe layer, get accelerated in it by an additional electric field, emitting proportional scintillation photons in the process. This gives the "$S2$" signal, delayed by the drift time. Since the drift velocity is constant for given electric field, the delay gives the $z$ position of the DM scattering event, while the $x$ and $y$ coordinates are determined by the position of the hits in the PMTs. Furthermore, the $S2/S1$ ratio is an important discriminator of DM signal over backgrounds, since nuclear recoils and electron recoils

have very different $S2/S1$ values, $(S2/S1)_{\mathrm{nr}} \ll (S2/S1)_{\mathrm{ee}}$. The 3D positioning of the scattering events allows to cut out the events close to the boundary of the detector, utilising the self-shielding property of Xe (in LXe the ambient radiation does not propagate very far). The recent two phase TPC experiments are LUX [326], PandaX-4T [186], XENON1T [185] and XENONnT [343] for Xe, and DarkSide-50 [325], ArDM [348] for Ar. The single phase experiments, XMASS using LXe [321], and DEAP-3600 using LAr [334], only measure the scintillation signal $S1$. These detectors benefit from a simpler design since there is no need for external electric fields, but the rejection against the backgrounds is harder. Therefore, all the upcoming and planned LXe and LAr experiments are of the dual phase type.

The dual phase design also allows to search for DM scattering or DM absorption on electrons, using the "$S2$ signal only" searches, at the cost of larger backgrounds. Similar types of analysis also allow for sub-GeV DM scattering on nucleons through the Migdal effect, see section 5.4.

### 5.6.4   Superheated liquids

The detectors based on superheated liquids are "yes or no" types of detectors [349]. The largest such detectors are operated as bubble chambers. A small deposit of energy due to DM scattering on a nucleus triggers a phase transition, creating a bubble of gas, but only if enough energy is deposited locally. A commonly used material is $C_3F_8$ or other liquids that contain fluorine. The most common fluorine isotope, $^{19}F$, has an unpaired proton. Thus, there is an enhanced sensitivity to spin-dependent DM/nucleus interactions due to a large corresponding hadronic matrix element.

The readout is in two channels: optical (searching for visible bubbles) and acoustic (measuring the pressure waves from the collapse of the bubbles). In order for bubbles to be created, the ionization density needs to be above a certain threshold. The DM interactions, neutron interactions or passing $\alpha$ particles ionize above the threshold, while the electromagnetic backgrounds, $\beta$ particles and $\gamma$'s, are below the threshold. Acoustic discrimination can be used to reject the $\alpha$ particle background, since the corresponding signal is louder than the one from DM scattering. The target volume is also monitored visually with digital cameras, detecting the formation of bubbles and their positions. In this way one can reject the surface events, which are more likely to be backgrounds.

Several different versions of detectors based on superheated liquids have been developed. In the past, the experiments with bubble chambers, described above, were the domain of the COUPP collaboration [350]. A different approach, based on superheated droplets embedded in a gel structure, was developed by SIMPLE [351] and PICASSO [352] collaborations. These types of detectors are more stable and cheaper, but have the downside that only a small part, a few percent, of the detector mass is the target material. The PICASSO and COUPP collaborations merged into PICO collaboration that is operating in SNOLAB a 60 kg bubble chamber of $C_3F_8$ [353,354]. A 40 l chamber is under construction, while in the future a 500 l chamber is planned, see table 5.4. A new concept for the bubble chamber, based on the "Geyser" technique, where the vapour above the liquid is kept at a lower temperature, is being developed by the MOSCAB collaboration [355]. Such detectors have, unlike the conventional bubble chambers, a very short reset time of a few seconds since the bubbles float to the top of the vessel, and then exit the liquid. In contrast, in the conventional bubble chambers one gets rid of the bubbles by increasing the pressure, which is a longer process.

### 5.6.5   The irreducible neutrino background

Coherent neutrino/nucleus scattering is an irreducible background to direct DM searches [221]. The neutrino/nucleus cross section for neutrinos with energy $E_\nu \lesssim$ GeV is approximately given by

$$\sigma_{\nu A} \approx \frac{G_{\mathrm{F}}^2}{4\pi} N^2 E_\nu^2 F^2(q), \tag{5.106}$$

where $N$ is the number of neutrons in the nucleus, and $F(q) \approx 1$ is the nuclear form factor for the SI scattering, cf. eq. (5.6). The neutrino/electron scattering cross section is similarly

$$\sigma_{\nu e} \approx \frac{G_{\mathrm{F}}^2}{4\pi} Z^2 E_\nu^2. \tag{5.107}$$

The solar neutrinos are the dominant background to DM–nucleus scattering for DM with a mass below $M \lesssim 10$ GeV, and for DM scattering on electrons, while the atmospheric and diffuse supernova background neutrinos are important for heavier DM masses, see fig.s 5.5, 5.6, 5.12. In the greyed out regions in the figures the neutrino background is important, and leads to the expected bounds on DM scattering to scale as $1/(\text{exposure})^{1/2}$, instead of the $1/(\text{exposure})$ scaling in the absence of backgrounds.

A DM signal due to SI scattering on a nucleus can be almost perfectly mimicked by the solar neutrino scattering, so that this background is sometimes referred to as the *"neutrino floor"* or the *"neutrino fog"*. The solar neutrinos give recoil energies, $E_R < 2E_\nu^2/m_A$, where $E_\nu$ is in the MeV regime, resulting in $E_R$ smaller than $\sim (4 \div 40)$ keV, with the precise number depending on the mass of the nucleus. The atmospheric neutrinos, created by cosmic ray collisions with the atmosphere, on the other hand, have a lower flux but reach higher energies, such that they will become a problem for heavy DM masses when the bounds reach $\sigma_{\mathrm{SI}}/M \lesssim 10^{-48}\,\mathrm{cm}^2/\,\mathrm{TeV}$. Amusingly, the background due to such neutrinos would be smaller for a detector on the moon, which has almost no atmosphere, while cosmic ray collisions with the lunar surface and crust lead to a less energetic neutrino flux [356].

An important discriminator against the neutrino background could be measurements done with different nuclei. While coherent neutrino scattering is dominated by couplings to neutrons, DM could couple to neutrons and protons with comparable strengths, or even more preferentially to protons, giving differing signals for scatterings on light versus heavy nuclei. This strategy is especially powerful, if DM/nucleon interactions are dominated by the spin-dependent scattering, the rate for which is highly dependent on the type of target used in the experiment.

Another possibility is to develop directional detectors, which we discuss next. In this case one could cut on events originating from the Sun's direction, thus reducing the solar neutrino background.

## 5.6.6 Directional detectors

The Earth is moving through the DM halo toward the Cygnus constellation (with Galactic coordinates $\ell \approx 90°$ and $b \approx 0°$), see section 2.3.2. In the Earth's rest frame there is therefore an apparent DM wind blowing from the direction of the Cygnus constellation. Since the DM/nucleus scattering peaks in the forward direction, the nuclear recoils also preferentially originate from the direction of the Cygnus constellation, but with a wider angular spread. The directionality of the DM signal can be a discriminator against the otherwise irreducible backgrounds that will be reached in the future, such as the events induced by the solar and atmospheric neutrinos (the so called neutrino fog, see section 5.6.5). A significant effort is therefore devoted toward developing directional detectors, which may be needed to go below the neutrino fog [357]. The detectors can have 1D capabilities, i.e., only distinguishing the forward scattering from the backward scattering, or the more powerful 2D or the full 3D reconstruction of the nuclear recoil track. In addition, if the information on the progression of the track is available (such as timing or the shape of the tracks), this allows for the head-tail discrimination, i.e., which direction did the track originate from. The signal in directional detectors carries enough information that one could simultaneously determine the DM mass, the scattering cross section as well as the DM velocity profile from a single 3D directional detector.

Several types of detectors that have potential directional sensitivity are being developed. Broadly speaking there are two approaches:

**Direct imaging of the recoil track**

Gas time projection chambers (TPCs) are the most mature technology. The gas needs to be at low pressure so that the keV-scale nuclear recoils result in $\sim$ mm ionization tracks, which can then be imaged with fine enough readout approaches. Because of the required low pressure the challenge for the gas TPC is always the small total target mass. The benefit is, beyond reducing backgrounds due to directionality of the signal, that many different types of gases or mixtures of gases can be used: Ne, $CH_4$, $CF_4$, $CHF_3$, $C_4H_{10}$,..., and thus can be sensitive to different DM interactions. The start of the research dates all the way to early 1990s [358], and is now a global effort, with DRIFT at Boulby mine, UK [315], MIMAC in Modane, France [359], NEWAGE in Kamioka, Japan [307], DMTPC at MIT, USA [316], $D^3$ at U. Hawaii and LBNL, USA [360], and CYGNO at Gran Sasso, Italy [361]. Recently, these groups formed a CYGNUS proto-collaboration, which is exploring a possibility of a ton-scale 'recoil observatory' [362]. This would reach the neutrino fog for GeV DM masses and potentially push below it. Other, less well developed ideas for DM detectors with directional capabilities are based on emulsions [363], crystal defects [364], 2D materials [365, 366] and using DNA strands [367]. The distribution of tracks could potentially also be used to search for a weak scale DM signal in the "paleo-detectors", i.e., the deep underground mineral rock samples with Gyr exposures (scanning of mica slabs was already used to set limits on ultraheavy DM, which would deposit the easier to distinguish long tracks) [368]. In general, in emulsions and solids there is a significant spread in the movement of the recoiled nucleus in the transverse direction, while in the low pressure gas environment of TPCs the directionality of the recoiled nucleus is much better preserved. Consequently, most of the research and development has focused on various versions of the gas TPCs.

**Indirect directional detection**

Even if the tracks from the individual DM scattering are not reconstructed, one can still gain some directional information on a statistical basis, if the response of the detector is anisotropic [369–371]. Anisotropic scintillators, such as $ZnWO_4$, have a light yield that depends on the recoil direction relative to the crystal axes [369]. This can then be correlated with the expected direction of the DM wind, taking into account both the annual and daily modulation, in order to increase the significance of the DM scattering signal. Interestingly, there is also some asymmetry in the ionization and scintillation responses of liquid argon (and potentially xenon) detectors due to the applied electric field, though the asymmetry is probably too small to significantly aid DM searches [371].

## 5.6.7   Searches for scattering on electrons

In any material one can have scattering of DM on either nuclei or on electrons, with the relative rates depending on the couplings of DM to electrons or to quarks and gluons. Both types of scattering deposit energy in the detector so in principle one can search for both using a single detector. However, the two types of scattering lead to different signals in detectors, so the question is whether experimentally the resulting signal can be picked up, and not be overwhelmed by the backgrounds. The experiments that have searched for DM scattering on electrons fall into two categories.

In the first group are the experiments that were primarily built to search for DM scattering on nuclei, and would typically treat electron recoils as backgrounds. Even so, lately many were able to place limits on DM/electron scattering by modifying their analyses. The prime examples are dual phase xenon detectors, in which in the nominal analysis the electron recoil events are treated as backgrounds and discarded, while in the $S2$-only analysis these events can be used to search for an excess due to DM scattering on electrons.

The second group consists of the experiments specifically built to search for DM/electron scattering. The prime examples are the SENSEI [342] and DAMIC [317] experiments, which use versions of silicon

| Experiment | Location | Operation | Technology | Target Mass | Home | Ref. |
|---|---|---|---|---|---|---|
| CDMS II | Soudan, MN | 2006 → 2007 | ath. phon.+ioniz. (50 mK) { | 4.6 kg Ge / 1.2 kg Si | web | [300] |
| XENON10 | Gran Sasso, Italy | 2006 → 2007 | scint.+ioniz. (179 K) | 5.4 kg Xe | web | [306] |
| EDELWEISS II | Modane, France | 2009 → 2010 | ther. phon.+ioniz. (18 mK) | 4 kg Ge | web | [310] |
| KIMS | Yangyang, S. Korea | 2009 → 2012 | scint. ($\sim$ 300 K) | 103.4 kg CsI(Tl) | web | [302] |
| EXO-200 | Carlsbad, NM | 2011 → 2018 | scint.+ioniz. ($\sim$ 167 K) | 82 kg Xe | web | [372] |
| CDMSLITE | Soudan, MN | 2012 → 2014 | ath. phon.+ioniz. ($\sim$ 50 mK) | 0.6 kg Ge | web | [322] |
| XMASS-I | Kamioka, Japan | 2013 → 2017 | scint. (173 K) | 832 kg Xe | web | [321] |
| DARKSIDE-50 | Gran Sasso, Italy | 2013 → 2018 | scint.+ioniz. ($\sim$ 80 K) | 46 kg Ar | web | [325] |
| LUX | Sanford, SD | 2013 | scint.+ioniz. ($\sim$ 180 K) | 118 kg Xe | web | [326] |
| EDELWEISS-III | Modane, France | 2014 → 2015 | ther. phon.+ioniz. (18 mK) | $\sim$ 16.5 kg Ge | web | [329] |
| CDEX-1 | Jinping, China | 2014 → 2017 | ioniz. ($\sim$ 90 K) | 0.9 kg Ge | web | [320] |
| PANDAX-II | JinPing, China | 2015 → 2018 | ioniz.+scint. ($\sim$ 165 K) | $\sim$ 117 kg Xe | web | [331] |
| DAMIC | SNOLAB, Canada | 2016 → 2017 | ioniz. ($\sim$ 105 − 140 K) | 6 → 40 g Si | web | [317] |
| SENSEI { | Fermilab, IL / SNOLAB, Canada | 2017 → 2020 / 2022 → | ioniz. (130 K) / ioniz. (130 K) | 95 mg → 1.9 g Si / $\sim$ 100 g Si | web | [342] |
| SUPERCDMS-HVEV { | Stanford, CA / Northwestern, IL | 2017 / 2019 | ath. phon. (36 mK) / ath. phon. (36 mK) | 1 g Si / 1 g Si | web | [373] |
| CDEX-10 | Jinping, China | 2017 → 2018 | ioniz. ($\sim$ 90 K) | 0.9 kg Ge | web | [338] |
| XENON1T | Gran Sasso, Italy | 2017 → 2018 | ioniz.+scint. (177 K) | 1042 kg Xe | web | [185] |
| BLANCO ET AL. | Chicago, USA | 2018 | scint. (274 K) | 1.3 kg $C_8H_{10}$ | paper | [374] |
| HOCHBERG ET AL. | MIT, USA | 2018 | supercond. nanowire (300 mK) | 4.3 ng WSi | paper | [222] |
| EDELWEISS-SUBGEV | Modane, France | 2019 | ther. phon. (20 mK) | 33.4 g Ge | web | [339] |
| LAMPOST | MIT, USA | 2021 | supercond. nanowire (300 mK) | $\mathcal{O}$(4 ng) WSi | paper | [232] |
| PANDAX-4T | Jinping, China | 2021 → | ioniz.+scint. ($\sim$ 170 K) | 2.7 t Xe | web | [186] |
| XENONNT | Gran Sasso, Italy | 2021 → | ioniz.+scint. (177 K) | 4.4 t Xe | web | [343] |
| DAMIC-M LBC | Modane, France | 2022 → | ioniz. ($\sim$ 130 K) | 8 g Si | web | [375] |
| JWST NIRSPEC | Lagrange point L2 | 2022 → | ioniz. (43 K) | 21 kg $Hg_{0.7}Cd_{0.3}Te$ | web | [376] |
| DAMIC-M | Modane, France | 2024 → | ioniz. ($\sim$ 120 K) | 26.4 g Si | web | [396] |

Table 5.3: *The past and present **direct detection experiments** bounding **scattering and/or absorption on electrons**. Experiments are ordered by the year of the first relevant physics run (3rd column), with the 2nd column showing the location, the 4th column the technology used and the 5th column the fiducial target mass. The last two columns give references for each experiment. The abbreviations used are the same as in table 5.1. Planned experiments are listed in table 5.4.*

CCDs (i.e., improved versions of the CCD sensors that are in everyday digital cameras) as ionization detectors. The energy transfer from DM to the detector is via electron recoil, creating a hole-electron pair, where the hole is captured in a potential well that can be read out repeatedly. This technique leads to very low energy thresholds all the way down to the Si band gap of 1.2 eV, since one can measure just a single hole–electron creation. This translates to a sensitivity to DM masses down to $\sim$MeV for DM/electron scattering, and to $\sim$eV for DM absorption. DAMIC and SENSEI are not directly sensitive to nuclear recoils in DM/nucleon scattering. However, the hole-electron creation can also be interpreted in terms of a Migdal effect in the DM/nucleon scattering, see section 5.4. The energy threshold can potentially be lowered by using superconducting nanowires as the target material, which then acts as a sensitive bolometer, changing from superconducting to normal conductor even if very small energies are deposited (the superconducting gap is only $\sim$ meV). The first bounds on DM/electron scattering [222] and dark photon absorption [232] using superconducting wires were placed already using prototype detectors, presently with comparable reach than the other experiments.

The experiments that placed limits on DM/electron scattering and/or DM absorption on electrons are listed in table 5.3.

### 5.6.8  Searches for neutrinos from the Sun and the Earth

The capture of DM particles in the center of celestial bodies such as the Sun and the Earth can lead to a signal in terms of high energy neutrinos. As discussed in more detail in section 6.9, the flux of neutrinos from the Sun due to DM annihilations is controlled by the DM capture rate, since this is the bottle neck. The bounds on neutrino flux from the Sun at energies above tens of MeV (which is high enough that the SM solar neutrino flux is negligible) would then impose constraints on DM capture rate, and by extension on the DM/proton scattering cross section, the same quantity that is probed by the conventional direct detection searches discussed above. The neutrino telescopes have higher thresholds, and therefore place bounds only for GeV DM masses and above.

Several neutrino telescopes derived such bounds. For annihilations in the center of the Sun (for SI and SD scattering): ANTARES [377], ICECUBE [194, 378, 379], KM3NET [380], BAKSAN [381], and SUPERKAMIOKANDE [382]. For capture and annihilations in the Earth (SI): ANTARES [383], ICECUBE [384], SUPERKAMIOKANDE [385]. A selection of these constraints is reported in the plots on direct detection bounds (SI, figure 5.5, and SD, figure 5.6). The bottom line is that, at the current stage, the constraints from the Sun on the SI cross section are about two orders of magnitude weaker than direct detection constraints. Constraints on the SD cross section are instead stronger than the direct detection constraints by about one order of magnitude (for the leptonic and gauge boson channels). Future HYPERKAMIOKANDE data are expected to improve constraints from the Sun by a factor of 2-3 [386]. On the other hand, given the current bounds from direct detection, the flux from the center of the Earth is predicted to be too weak to give a relevant signal.

## 5.7  Direct detection: status

The current most stringent bounds on SI DM/nucleus scattering cross sections for various target materials are shown in fig. 5.5, with bounds on SD scattering cross sections shown in fig. 5.6, while tables 5.1 and 5.2 contain the complete list of experiments that have placed limits on such scatterings. Similarly, fig. 5.12 shows the current bounds on SI DM/electron scattering cross section, while table 5.3 lists experiments that have set bounds on DM/electron scattering or on DM absorption on electrons. Note that using annual modulation the DM-electron scattering signal could also be searched for in neutrino detectors such as JUNO [406], though typically with smaller sensitivity than the dedicated experiments, and we thus do not list them in the table. Note also that the shown constraints use the value of the local DM density $\rho_\odot = 0.40$ GeV/cm$^3$, see eq. (2.11), which differs from the commonly adopted value in the direct dark matter detection literature $\rho_\odot = 0.30$ GeV/cm$^3$ [407]. In the exclusion plots that are focused on relatively small cross sections, fig. 5.5 (bottom), fig. 5.6 and fig. 5.12, we only show the lower bounds on the excluded regions not to clutter the figures futher. We thus do not denote explicitly at how large values of the cross sections the exclusions from each experiments are no longer applicable, since in that case DM is no longer able to reach the underground detectors (this upper boundaries are denoted for SI scattering on nuclei in fig. 5.5 (top)). In general, such large cross sections are excluded from other searches, but for electron scattering via a light mediator there is possibly still a very small allowed region for $\sigma_e \sim \mathcal{O}(10^{-21})$ cm$^2$ and DM mass around GeV [408]. Finally, table 5.4 lists planned direct detection experiments, as well as concept and R&D projects.

## 5.8  Laboratory searches for light and ultra-light DM

As discussed in section 3.4, DM could be a field lighter than about an eV. If this is the case, in order to match the observed DM relic abundance, DM quanta must have large occupancy numbers, $N \gg 1$, i.e., with many DM particles overlapping within a single Compton wavelength. Light or ultra-light DM is

| Experiment | Location | Data Taking | Readout | Target | Home | Ref. |
|---|---|---|---|---|---|---|
| DARKSIDE-20K | Gran Sasso, Italy | 2026 | scint.+ioniz. ($\sim$ 85 K) | 20 t Ar | web | [387] |
| DARKSIDE-LowMass | Gran Sasso, Italy | | scint.+ioniz. ($\sim$ 85 K) | $\sim$1 t Ar | web | [387] |
| SBC | SNOLAB, Canada | 2028 | scint. bubble chamb. ($\sim$ 100 K) | 10 kg Ar | talk | [388] |
| ARGO | SNOLAB, Canada | 2029 | scint.+ioniz. ($\sim$ 85 K) | 300 t Ar | web | web |
| DARKSIDE-LM | | | scint.+ioniz. ($\sim$ 85 K) | 1.5 t Ar | web | [389] |
| LZ-HYDROX | Sanford, SD | 202x | ioniz.+scint. (174 K) | 5.5 t Xe + 2 kg $H_2$ | web | LOI |
| XLZD | undetermined | 203x | scint.+ioniz. ($\sim$ 170 K) | 60/80 t Xe | web | [347] |
| PANDAX-20T | Jinping, China | 2027 | scint.+ioniz. ($\sim$ 170 K) | $\sim$ 20 t Xe | web | talk |
| PANDAX-xT | Jinping, China | 203x | scint.+ioniz. ($\sim$ 170 K) | $\sim$ 40 t Xe | web | [390] |
| QUEST-DMC | | | quasipart. ($\sim$ 100 $\mu$K) | 1 cm$^3$ $^3$He | paper | [391] |
| DELIGHT | Vue-des-Alpes, Switz. | 202x | phon.+roton ($\sim$ 20 mK) | 10 l $^4$He | web | [392] |
| HeRALD | | 202x | phon.+roton ($\sim$ 50 mK) | $\sim$ 1 kg $^4$He | web | [393] |
| SUPERCDMS SNOLAB | SNOLAB, Canada | 2023$\rightarrow$ { | ath. phon.[+ioniz.] (15 mK) / ath. phon.[+ioniz.] (15 mK) | 11[+14] kg Ge / 2.4[+1.2] kg Si | web | [394] |
| RES-NOVA | Gran Sasso, Italy | | ther. phon.+ioniz. | $\sim$ 170 kg $\rightarrow$ 1.8 t PbWO$_4$ | paper | [395] |
| OSCURA | SNOLAB, Canada | 2029 | ioniz. ($\sim$ 130 K) | 10 kg Si | web | [397] |
| CDEX-50 | Jinping, China | 202x | ioniz. ($\sim$ 90 K) | $\sim$ 300 kg Ge | web | talk |
| EDELWEISS-CRYOSEL | Modane, France | 202x | ath. phon. ($\sim$ 10 mK) | $\sim$ 30 g Ge | web | [398] |
| CDEX-300 | Jinping, China | 2027 | ioniz. ($\sim$ 90 K) | $\sim$ 300 kg Ge | web | LOI |
| CDEX-1T | Jinping, China | 2033 | ioniz. ($\sim$ 90 K) | $\sim$ 1 t Ge | web | LOI |
| CDEX-10T | Jinping, China | 2040 | ioniz. ($\sim$ 90 K) | $\sim$ 10 t Ge | web | LOI |
| COSINE-200 | Yemilab, South Korea | 2024$\rightarrow$ | scint. ($\sim$ 300 K) | $\sim$ 200 kg NaI(Tl) | web | talk |
| COSINE-100U | Yemilab, South Korea | 2025 | scint. ($\sim$ 300 K) | 99.1 kg NaI(Tl) | web | [399] |
| COSINUS | Gran Sasso, Italy | 2025 | scint. ($\sim$ 10 mK) | 0.24 $\rightarrow\sim$ 1 kg NaI(Tl) | web | [400] |
| SABRE { | Gran Sasso, Italy | 2025 | scint. ($\sim$ 300 K) | 50 kg NaI(Tl) | web | [345] |
| | SUPL, Australia | 2026 | scint. ($\sim$ 300 K) | 50 kg NaI(Tl) | web | |
| PICOLON | Kamioka, Japan | 202x | scint. ($\sim$ 300 K) | 54 $\rightarrow$ 250 kg NaI(Tl) | paper | [401] |
| KamLAND-PICO | Kamioka, Japan | 203x | scint. ($\sim$ 300 K) | 1000 kg NaI(Tl) | paper | [401] |
| DMICE-250 | South Pole | | scint. ($\sim$ 260 K) | $\sim$ 200 kg NaI(Tl) | talk | talk |
| PICO-40L | SNOLAB, Canada | 2025 | bubble chamber ($\sim$ 290 K) | $\sim$ 50 kg C$_3$F$_8$ | web | [402] |
| PICO-500 | SNOLAB, Canada | 2027 | bubble chamber ($\sim$ 290 K) | 360 kg C$_3$F$_8$ | web | [403] |
| MOSCAB | Gran Sasso, Italy | 202x | bubble chamber ($\sim$ 290 K) | 2 $\rightarrow$ 25 l C$_3$F$_8$ | paper | [355] |
| MIMAC | Grenoble, France | | ioniz. ($\sim$ 300 K) | CF$_4$+CHF$_3$ | paper | [359] |
| NEWS-G : ECUME | SNOLAB, Canada | | ioniz. ($\sim$ 300 K) | $\sim$ 2 kg CH$_4$ | web | [340] |
| NEWS-G : DARKSPHERE | Boulby, UK | | ioniz. ($\sim$ 300 K) | 27 kg He+C$_4$H$_{10}$ | web | [340] |
| CYGNO | Gran Sasso, Italy | 2024$\rightarrow$ | ioniz. ($\sim$ 300 K) | 1 m$^3$ He+CF$_4$ | web | [361] |
| CYGNUS | multiple sites | | ioniz. ($\sim$ 300 K) | 10$^3$ m$^3$ He+SF$_6$/CF$_4$ | web | [362] |
| SNOWBALL | | | supercooled liq. ($\sim$ 250 K) | 1 kg H$_2$O | talk | [404] |
| ALETHEA | | | scint.+ioniz. ($\sim$ 4 K) | 10 kg He | paper | [405] |
| SPLENDOR | | | ioniz | Eu$_5$In$_2$Sb$_6$, EuZn$_2$P$_2$ | poster | LOI |
| TESSERACT | Modane, France | 2028$\rightarrow$ | ath. phon. | Al$_2$O$_3$, GaAs, He | web | LOI |
| WINDCHIME | | | accelerometers | | paper | [271] |

Table 5.4: **Planned future, as well as concept and R&D projects for direct detection experiments.** *Experiments are grouped by the target material, and ordered by the year of the expected first physics run (3rd column), with the 2nd column showing the location, the 4th column the technology used and the 5th column the fiducial target mass, with the last two columns listing the references. The abbreviations used are the same as in table 5.1, and in addition: "supercooled liq." $\rightarrow$ "supercooled liquid", "scint. bubble chamb." $\rightarrow$ "scintillating bubble chamber".*

thus necessarily bosonic (either a scalar, $\varphi$, a pseudoscalar, $a$, or a vector, $V_\mu$) and behaves as a classical oscillating field. Such light DM can be searched for with cosmological and astrophysical probes, as well as with laboratory experiments. Here we focus on the latter, while the astrophysical and cosmological probes are discussed in section 6.14.

The DM field would oscillate as

$$\phi(t) = \phi_0 \cos(Mt + \beta), \qquad (5.108)$$

where $\beta$ is a phase. Assuming a nearly free field, the amplitude is

$$\phi_0 = \sqrt{2\rho}/M, \qquad (5.109)$$

with $\rho$ the local DM density. For laboratory experiments this is the DM density in the solar system, $\rho_\odot$, see section 2.2.2. DM particles have a small energy spread, $\Delta E_\phi$, so that the classical field in eq. (5.108) has a long phase coherence time $\tau_{\rm coh} \approx 2\pi/\Delta E_\phi$. For DM particles gravitationally bound in the Milky Way, $\Delta E_\phi$ is expected to be comparable to the kinetic energy due to orbiting around the Galactic center. This gives the estimate $\tau_{\rm coh} \approx 4\pi/m_\phi\langle v^2 \rangle \sim 10^6\,\tau_{\rm osc}$, where $\tau_{\rm osc} = 2\pi/m_\phi$ is the oscillation time, and we used that $\langle v^2 \rangle^{1/2} \sim 10^{-3}$ at the position of the solar system. That is, the oscillations of the DM classical field are coherent over many oscillations.

A number of experimental techniques can exploit this insight to search for light or ultralight DM [409]:

◇ **Atomic, molecular, and nuclear clocks**. The oscillating DM field can induce time-dependent changes in the electromagnetic constant, $\alpha$, and in the fermion masses, $m_f$. These time-dependent variations can be searched for using atomic clocks. For instance, a light scalar field $\phi$ coupling linearly to the SM fields through the effective Lagrangian[24]

$$\mathscr{L}_{\rm eff} = \frac{g_\gamma}{4}\phi F_{\mu\nu}F^{\mu\nu} - \sum_{f=e,p,n,} g_f \phi \bar{f}f, \tag{5.110}$$

induces the following changes

$$\alpha \to \frac{\alpha}{1 - g_\gamma\phi} \simeq \alpha\big(1 + g_\gamma\phi\big), \qquad m_f \to m_f + g_f\phi. \tag{5.111}$$

That is, both $\alpha$ and $m_f$ become time dependent, oscillating with frequency $M$, and the amplitude of oscillations is $g_{\gamma,f}\sqrt{2\rho}/M$.

The $\phi$ field might instead couple quadratically to the SM, e.g., if it is odd under a $\mathbb{Z}_2$. The interaction Lagrangian is then obtained by replacing $\phi \to \phi^2$ in eq.s (5.110), (5.111). This results in oscillations of $\alpha$ and $m_f$ with frequency $2M$, since $\cos^2(Mt) = (1 + \cos 2Mt)/2$. There is also a time-independent shift in $\alpha$ and $m_f$ that depends on the value of DM density and induces variations in fundamental constants that are correlated with differences in local DM densities.

The space and time variations of $\alpha$ and of $m_e/m_{p,n}$ ratios translate to variations in atomic, molecular and nuclear spectra, and thus in changes of frequencies of clocks that use these various transitions. The optical clocks based on atomic transitions are mostly sensitive to a variation in $\alpha$. The microwave clocks, on the other hand, are based on transitions between hyperfine split states, and are thus sensitive to a combination of $\alpha$ and $m_e/m_p$. Both of these types of clocks are exceedingly stable and now reach relative precision on transition frequencies below $10^{-18}$. Since optical and microwave clock frequencies are shifted differently, one can then search for the imprints of oscillating light or ultralight DM field by monitoring for phase shifts between the two types of clocks. Typically such searches are less sensitive than other constraints, but the field is quickly developing, with possibly world leading future sensitivities in the range of ultralight DM masses $m_\phi \approx 10^{-10} - 10^{-6}\,{\rm eV}$.

◇ **Atom interferometers.** An atom interferometer compares the phases accumulated by two clouds of cooled atoms, displaced by a macroscopic distance $L$. The separation can range from $\sim 1$ meter (in current experiments [410]), to few $100\,{\rm m}$ (in planned experiments [411]). The atoms are manipulated by a series of laser pulses such that their wave functions are in a coherent superposition of a ground and an excited state. Their energy levels differ by $\Delta\omega_A$, and thus the time evolution introduces a relative phase difference $\exp(i\int dt\,\Delta\omega_A)$. The phase difference imprints itself in the final interference pattern of the two atomic clouds, i.e., the number of atoms in the excited vs.

---

[24]We work at low energies, below the QCD confinement scale, and thus the relevant degrees of freedoms are photons, electrons, protons and neutrons.

ground state. Many experimental uncertainties cancel when comparing the interference patterns in the two clouds. For instance, the same laser beam can be used to manipulate both clouds of cooled atoms, and so a high stability of laser phase is less important than for atomic clocks. Furthermore, the atomic clouds are launched upwards and fall freely under gravity during measurements, and are thus shielded from environmental noise.

The passing of laser beams sets the time for transitions between ground and excited states, which are then shifted for the two clouds by the time it takes for the light to travel between them. The differences in the interference patterns are therefore controlled by an interferometer phase proportional to the separation, $\delta\phi \propto \Delta\omega_A L/c$. Couplings of DM to SM particles induce a time-dependent $\Delta\omega_A$, which can be searched for experimentally in the interference patterns [412].[25]

◇ **Optical cavities and mechanical resonators.** The changes in $\alpha$ and $m_e$ translate into a changed length of solid objects due to the change in the size of the electron clouds surrounding the nuclei in the material [413]. The sizes of the atoms are governed by the Bohr radius, $a_{\mathrm{Bohr}} = 1/m_e\alpha$, and thus the change in the length $L$ of a solid object is given by

$$\frac{\delta L}{L} \simeq \frac{\delta a_{\mathrm{Bohr}}}{a_{\mathrm{Bohr}}} = -\frac{\delta\alpha}{\alpha} - \frac{\delta m_e}{m_e}, \tag{5.112}$$

where $\delta\alpha$ and $\delta m_e$ are the time-dependent changes in eq. (5.111), due to the couplings of matter to the oscillating DM field. The changes in $\delta L$ are resonantly enhanced, if the oscillation frequency matches the vibrational mode of the solid, especially for resonators with large mechanical quality factors, such as optical cavities. The change of the linear dimensions of the cavity changes the resonant frequency of the cavity, $\nu_{\mathrm{cavity}} \propto 1/L$. This can be detected by comparing $\delta\nu_{\mathrm{cavity}}$ with the shift in a frequency of an atomic transition, each of which depends differently on $\delta\alpha$ and $\delta m_e$. The other option is to compare two cavities: one with rigidly attached sides, the distance between which then changes with DM frequency, and another with suspended sides, which do not.

◇ **Optical interferometers** are exquisitely sensitive to changes in the optical lengths in the two arms. This has been used to detect gravitational waves. Oscillating light DM gives a signal by changing the thickness of the beam splitter, resulting in the time-dependence of interference patterns [414].

◇ **Torsional balances** are used to search for non-gravitational forces between macroscopic test masses, by searching for small differential accelerations perpendicular to the torsion fiber. The light DM as a force carrier can be searched for either in the static limit, where Earth is used as a (large) source of DM induced potential, or in a time-dependent search, where one searches for oscillatory behaviour due to the new DM induced forces [416]. Such tests also go under the name of *5-th force searches* or *tests of the equivalence principle*.

◇ **Axion detectors**. Section 10.4 will show that the axion $a$ is a motivated light boson DM candidate that couples to photons via the $aF_{\mu\nu}\tilde{F}^{\mu\nu} = -4a\boldsymbol{E}\cdot\boldsymbol{B}$ effective operator. In the presence of an external magnetic field $\boldsymbol{B}$, the oscillating DM axion field acts as a new source of electric and magnetic fields, which can be used to construct a variety of possible detection techniques, such as shining-through wall-experiments, helioscopes and haloscopes, to be discussed in section 10.4.5. Many of these experiments can be used to also search for dark photons. Since the dark photons interact with SM through kinetic mixing, the signals are then present also in the absence of an external magnetic field, and can be searched for even more readily than the axions. A review of uses of axion detectors for dark photon searches can be found in [409].

---

[25]Another major scientific goal of atom interferometers is the search for gravitational waves, which would affect the distance $L$ between the clouds, and as a result leave an imprint in the interference patterns.

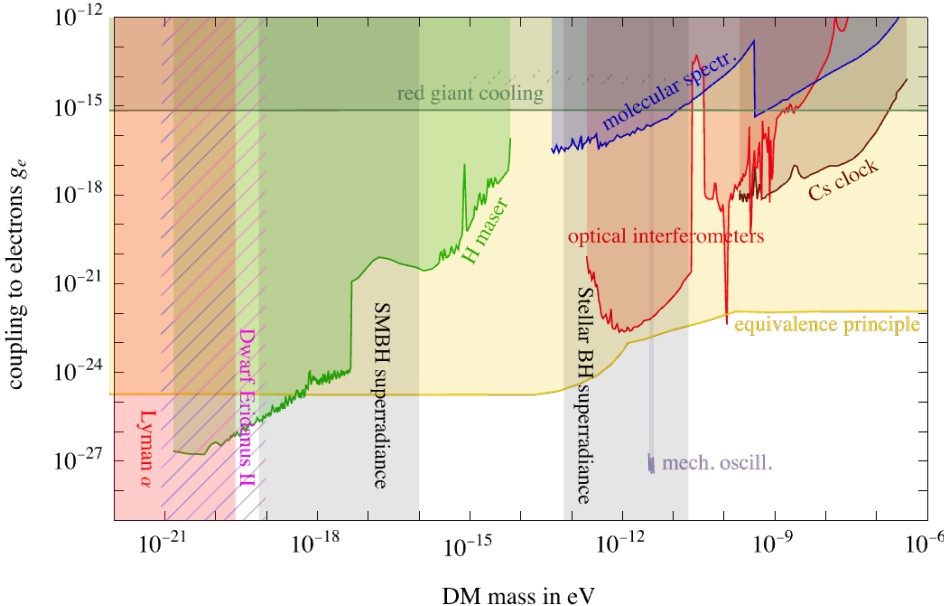

Figure 5.19: *Laboratory **constraints on light and ultralight scalar DM**, assumed to couple only to electrons ($g_e \neq 0$ in eq. (5.110)): from comparisons of H masers and cavities, molecular spectroscopy of $I_2$, optical interferometers, comparison of atomic Cs clock with reference cavity oscillator, and from tests of the equivalence principle [415]. There are also bounds from astrophysics (excluded vertical bands, see fig. 6.17 and section 6.14 for further details), as well as constraints from the absence of anomalously large cooling rates in red giants, which exclude the upper region of the shown range in $g_e$.*

◇ **Spin-based sensors**. Axions might also couple to SM fermions via the relativistic operator $\partial_\mu a(\bar{f}_i \gamma^\mu \gamma_5 f_i)/f_{a,i}$, that results in the non-relativistic Hamiltonian

$$\mathcal{H} = -\frac{2}{f_{a,i}} \boldsymbol{S}_i \cdot \boldsymbol{\nabla} a(\boldsymbol{r}, t). \tag{5.113}$$

The derivative coupling is a consequence of axion being a result of a spontaneously broken global symmetry (a (pseudo)Nambu-Goldstone boson), while the coupling to nuclear or electron spin, $\boldsymbol{S}_i$, is possible due to the fact that axion is a pseudoscalar. Vector DM would instead couple as $\mathcal{H} \propto \boldsymbol{\phi} \cdot \boldsymbol{S}_i$, without derivatives. Such interactions with DM would therefore appear as a new magnetic-like force acting on nuclear or electron spins, and can be searched for using spin-based quantum sensors [417, 418], see also section 10.4.5.

◇ **Neutrinos**. If DM $\phi$ has an effective Yukawa coupling, $\mathscr{L} \supset g_\nu \, \phi \nu_L^2/2 +$h.c., to SM neutrinos, their masses receive corrections $\delta m_\nu \approx g_\nu \phi \approx g_\nu \, \text{eV}(\text{meV}/M)$, where the numerical estimate assumes the local DM density and eq. (5.109). For light enough DM and big enough $g_\nu$ this coukd lead to a correction comparable or larger than the observed neutrino masses [419]. However, significant effects from such couplings in present day neutrino experiments are disfavoured by cosmology: since the DM density was higher in the earlier universe, CMB and BBN observations imply constraints $g_\nu \lesssim M/\,\text{eV}$. Effects in neutrino oscillations compatible with the observed cosmology are possible, if DM instead couples to relatively light right-handed neutrinos [419].

Fig. 5.19 shows an example of laboratory constraints on the electrophilic light/ultra-light scalar DM

(only $g_e$ taken to be nonzero). The bounds from comparisons of Cs clocks and H masers with other resonators (cavities), are typically less sensitive than the searches for violations of equivalence principle, which test for the presence of a new force mediated by a light scalar. See also fig. 6.17 and section 6.14 for the discussion of astrophysical bounds, and [409] for a more detailed discussion of constraints. Another example of a light scalar is the light Higgs-mixed scalar with constraints shown in fig. 7.4. Yet another example of a light boson is an axion, with the constraints on axion couplings to photons shown in fig. 10.1.

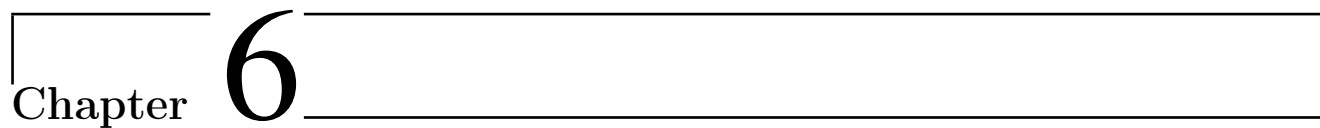

Chapter **6**

# Indirect detection

Indirect Detection (ID) experiments search for the signal illustrated in fig. 6.1: Dark Matter particles pair-annihilate (or decay) in the galactic halo or outside it, and this results in Standard Model particles that can be detected by looking for an excess in the cosmic ray fluxes measured on Earth (or near it), above the presumed astrophysical contribution. This chapter focuses mainly on the 'classic' indirect detection signals of DM particles in the GeV to TeV mass range, typically (but not necessarily) in the WIMP class. The same concepts, however, apply also to lower and higher DM masses, as well as to many different classes of DM candidates. We will thus try to keep most of the discussion as broad as possible.

The flux of cosmic rays that originates from DM is primarily determined by either the DM annihilation cross section or the DM decay rate. An important benchmark for the indirect detection searches is the DM annihilation cross section that leads to the observed cosmological DM abundance in the case of a thermal freeze-out, see eq. (4.13). The size of the expected signal also depends on other aspects of DM: its spatial distribution is particularly important. The signal rate tends to be higher from regions where DM is expected to be the densest, such as the center of our Galaxy, the inner halo of our Galaxy, nearby galaxies dominated by Dark Matter, the center of the Sun, the center of the Earth... However, some of these promising regions (notably the Galactic Center) are also the most complicated ones from the point of view of the underlying astrophysics. The best detection opportunities might therefore not come from the targets with the highest DM content, but rather from those with the most favourable signal over background ratio.

The best detection opportunity also depends on the specific cosmic ray species under investigation. For charged cosmic rays, for instance, it is typically better to focus on antimatter particles (positrons, anti-protons, anti-deuterons... ) since they are less likely to be produced in the astrophysical processes than the corresponding matter particles.[1] This explains why DM studies often concentrate on anti-protons rather than on more abundant protons: in typical DM models both are produced in equal quantities (in the absence of any model specific matter-antimatter violations), but the astrophysical background is smaller for anti-protons.

This brings us to the important role the astrophysical backgrounds play in indirect detection: the general strategy in indirect detection is to search for a 'bump' (an excess) in the spectrum of cosmic rays on top of a background that is assumed to be a power-law in the energy of incoming cosmic ray particles. This power law behaviour is motivated by the basic acceleration mechanisms of charged particles in astrophysical environments. In the real cases, however, the situation is often more complicated: one often encounters deviations from the power-law behaviour, spectral breaks, unexpected bump-like contributions from astrophysical sources,... Several examples that we discuss in detail in chapter 8 illustrate well how the searches work in practice.

---

[1]See, e.g., ref. [420] for comparative plots.

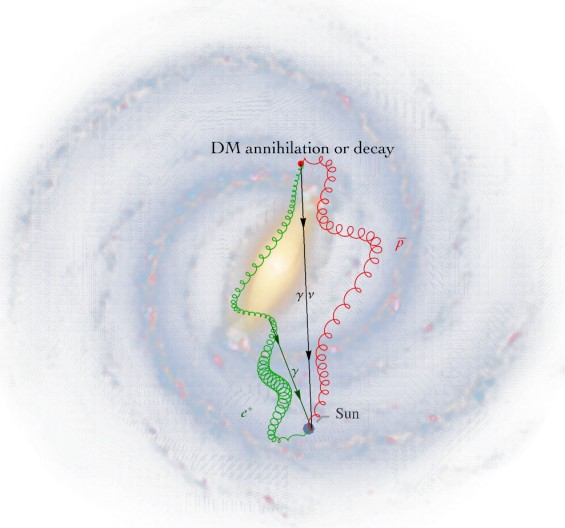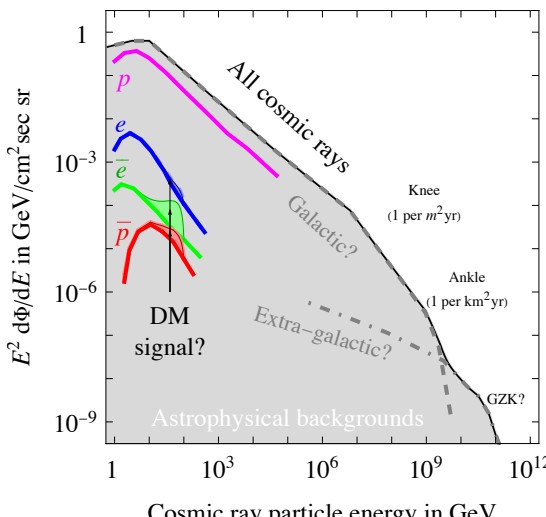

Figure 6.1: ***Cartoon of indirect detection***: *DM annihilations or decays in the Milky Way produce SM particles that reach the observer in a solar system, possibly producing an excess in the energy spectrum of cosmic rays over the observed astrophysical background.*

In general terms, the particles that are produced by DM annihilations/decays and that we hope to detect are all the stable Standard Model species: photons, neutrinos, positrons, anti-protons, anti-deuterium and maybe even more exotic anti-nuclei such as anti-helium (plus electrons, protons and light nuclei, which are however disfavored by their more abundant astrophysical backgrounds, as mentioned above). Each of these messengers possesses distinct advantages and disadvantages, when considered as DM signals:

○ **High-energy photons ($\gamma$-rays)** propagate freely in the galactic environment,[2] so that the information about DM lies both in their energy spectrum and in the angular distribution.[3] However, since DM is electrically neutral it typically produces photons only via subdominant mechanisms, e.g., via loops involving charged particles, or as an ancillary radiation. The photon flux is thus expected to be somewhat suppressed and often model-dependent. Since these photons are produced directly in the microphysical processes involving DM, with no interactions with the environment, they are referred to as the *prompt $\gamma$-rays*.

The typically considered sources are very diverse: the area around the galactic center, dwarf galaxies, other nearby galaxies, galaxy clusters or the entire distribution of DM in the Universe (see section 6.2).

○ **Low-energy photons (X-rays, radio waves)** can be produced by the electrons and positrons originating from DM, when these undergo additional interactions (e.g., photons emitted in inverse Compton scattering, synchrotron emission, or bremsstrahlung). Such photons constitute a DM signal, but are 'doubly indirect', since they depend on the astrophysical environment that reprocesses

---

[2]In the extragalactic/cosmological environment, on the other hand, absorption can occur. See section 6.2.3.

[3]Photon *polarization* has also been considered as possible signal of DM [421]. Photons from DM can exhibit a net circular polarization in specific models, such as the decaying asymmetric DM, DM annihilations that violate P and CP symmetries, DM interactions with ambient cosmic rays, etc. Detecting polarization of high energy photons is, however, very challenging experimentally.

the $e^\pm$. Because of this, there are additional uncertainties due to the strength of the magnetic field, the gas density, etc. The photons of this type are referred to as the *secondary radiation*.

The low energy photons are, however, not limited to just the secondary production. The $X$-rays and other low energy radiation can also arise directly from the decays of light DM particles with masses in the keV to MeV range, e.g., in models of DM as sterile neutrinos, or in the evaporation of PBH DM.

○ **Positrons** diffuse in the galactic magnetic fields, which randomizes their directions, so that they do not directly track back to the DM distribution. The information about DM therefore lies only in the energy spectrum. However, the latter is significantly affected by the fact that positrons, while propagating through the Galaxy, lose energy through various mechanisms such as synchrotron emission, Coulomb scattering, ionization, bremsstrahlung and Inverse Compton (IC) scattering processes. Furthermore, below a few GeV the solar activity distorts their spectrum, and below ∼1 GeV it deflects them away, so that they cannot even penetrate into the solar system. High-energy positrons coming from the nearby regions of the Galaxy are less affected by diffusion and losses, offering in principle a cleaner access to potential DM contributions to the flux.

○ **Electrons**. Similar comments as for positrons apply also to the DM produced electrons, with the disadvantage of a higher astrophysical background. The advantage, on the other hand, is that at high energies it is easier to measure the total $e^- + e^+$ flux rather than the positron flux alone.

○ **Anti-protons** diffuse in the galactic magnetic fields, which randomizes their directions. Unlike $e^\pm$, they undergo negligible energy losses, up to some scatterings on matter in the galactic plane. This means that even far-away regions of the Galaxy contribute to the anti-proton flux measured on Earth, and therefore its magnitude has significant astrophysical uncertainties. Because of the limited energy losses, the spectral shape is somewhat preserved and thus the information about DM lies also in it. However, compared to $e^\pm$ case, the anti-proton spectra for different DM annihilation/decay channels are more self-similar and thus carry less discriminating power (see section 6.1). The solar activity again distorts the spectrum below a few GeV and deflects anti-protons away from the solar system for energies below a fraction of a GeV.

○ **Anti-deuterons**. Nuclei of anti-deuterium can be synthesised via the coalescence of an anti-proton and an anti-neutron produced in the DM annihilation or decay process. The expected yield is very small. On the other hand, the astrophysical background is also expected to be small and, notably, is expected to peak in a range of energies different from the one typical of a DM signal, thanks to the differing kinematics of the two production mechanisms. The propagation of anti-deuterons in the galactic environment is analogous to anti-protons. Heavier anti-nuclei, such as anti-helium, can be produced in a completely analogous way, with the important penalty of a much more suppressed flux, due to the need for more anti-nucleons to coalesce.

○ **Neutrinos** propagate freely in the Galaxy and can also propagate through the dense matter of the Earth and of the Sun, up to multi-TeV energies. The small interaction cross sections make the detection of neutrinos more difficult than, e.g., of gamma rays. Furthermore, the neutrino energy can often be reconstructed only partially because they are measured indirectly via the detection of charged particles (e.g. up-going muons) produced by a neutrino interacting in the material of a neutrino telescope, or in the rock or water or ice surrounding it. On the other hand, the neutrino interaction cross section increases with energy, thus partly compensating the decrease in flux for larger DM masses. Possible sources of neutrinos from DM are the same as those for photons, but in addition, also the center of the Sun and, less promising, the Earth.

Pioneering works have proposed these different messengers as a promising avenue for DM discovery already starting in the late-1970's [422]. Gamma rays from annihilations were first considered in Gunn

et al. (1978), Stecker (1978), and Zeldovich et al. (1980),[4] and then revisited in Ellis et al. (1988). Antiprotons were first discussed in Silk & Srednicki (1984), Stecker et al. (1985) and then more systematically in Ellis et al. (1988), Rudaz & Stecker (1988), Stecker & Tylka (1989). Positrons were discussed in Silk & Srednicki (1984), Ellis et al. (1988), Rudaz & Stecker (1988), Turner & Wilczek (1990). Anti-deuterons have been first discussed in Donato et al. (2000 and 2008), Baer & Profumo (2005), Kadastik et al. (2010). The similar case of anti-helium was first considered in Carlson et al. (2015) and Cirelli et al. (2015). Radiowaves from synchrotron radiation from DM were first explored in Berezinsky et al. (1992), Berezinsky et al. (1994), Gondolo (2000), Bertone et al. (2001) and later in Aloisio et al. (2004). Cosmological gamma rays were first discussed in Gunn et al. (1978) and Stecker (1978), but then systematically reconsidered in Bergstrom et al. (2001) and Ullio et al. (2002). Inverse Compton gamma rays from DM have only been considered relatively recently as a possible signal (see e.g. Baltz & Wai (2004), Cholis et al. (2009), Zhang et al. (2009)), and similarly for bremsstrahlung gamma rays in Cirelli et al. (2013). Neutrinos from the center of the Sun and the Earth were first considered by Gould (1987) [424].

The first step in computing the indirect detection signals is to calculate the spectra of SM particles as they emerge from DM annihilation or decay, i.e., the spectra at production. These are discussed in section 6.1, in a model-independent way. The spectra at production then have to be convoluted with the information about the source (the distribution of DM, in the Galaxy or elsewhere, as well as the distributions of gas, light and magnetic fields for the secondary radiation) and on the propagation of the SM particles in the astrophysical environments. These aspects are covered in sections 6.2 to 6.8. More precisely, in section 6.2 we discuss the basic observables for prompt $\gamma$-rays, and in section 6.3 for neutrinos. In section 6.4 we discuss electrons and positrons, including, in particular, their propagation in the Galaxy, while in section 6.5 we discuss anti-proton fluxes and their propagation, and in section 6.6 a similar case of anti-deuterons and anti-helium. Section 6.7 is dedicated to the computation of fluxes of secondary radiation: inverse Compton $\gamma$-rays, bremsstrahlung $\gamma$-rays and synchrotron radiation. Section 6.8 discusses which modifications of the formalism are needed in order to take into account several possible sources of enhancements.

The rest of the chapter can be read in two different ways. On the one hand, after section 6.8 the reader might want to jump directly to section 6.13, where we review the current status of the searches in the field, mostly in terms of constraints on DM annihilations/decays. (The existing anomalies and possible signals of detection are covered in chapter 8.) Alternatively, one can continue with sections 6.9 to 6.12, which contain a variety of other indirect constraints on DM, before moving to section 6.13. Section 6.9 deals with neutrinos from the Sun or the Earth: they require a separate treatment that is distinct from our discussion of galactic cosmic rays, both in terms of spectra at production and in terms of the peculiar propagation in celestial bodies, including oscillations. Section 6.10 discusses DM annihilations/decays that occur inside astrophysical objects. Section 6.11 describes the impact of DM annihilations/decays during the so-called dark ages of cosmology, i.e., on the CMB and on reionization. Section 6.12 gives constraints on DM due to Big Bang Nucleosynthesis. Finally, section 6.14 reviews astrophysical searches for ultralight DM.

While, as already mentioned above, this chapter is mostly oriented toward the indirect detection of GeV to TeV particle DM [5], most of the discussion of the underlying physics remains relevant also for other ID signals, e.g., sub-GeV particle DM (its status is discussed in section 6.13.2), cosmic rays from evaporating Primordial Black Holes (see section 3.1.1), or $X$-rays from decaying sterile neutrino DM (see e.g. section 8.2.3).

---

[4]The specific case of gamma ray lines was first discussed in Srednicki, Theisen and Silk (1986) and in Bergstrom and Snellman (1988), in [423].

[5]For this range of masses, most of the tools and the results in this chapter can be reproduced with the PPPC4DMID [425].

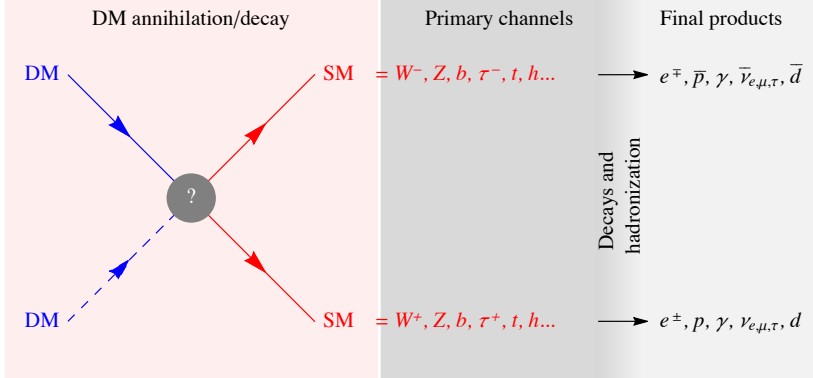

Figure 6.2:   ***DM Indirect Detection****: the general schematic, illustrating the generation of primary particle spectra.*

## 6.1   Energy spectra at production

DM particles pair-annihilate (or decay) in astrophysical environments, thereby creating SM particles that then undergo the processes of decay, showering and hadronization. The end result are fluxes of stable SM particles ($e^{\pm}, \bar{p}, \bar{d}, \gamma, {}^{(\overline{\nu})}{}_{e,\mu,\tau}$) that can be searched for in cosmic rays as a signal of DM. The whole process is succinctly represented in fig. 6.2.

   For many phenomenological analyses, it is useful to compute such fluxes in a model independent way: in the first step one assumes that the DM annihilations or decays produce a SM final state that is a particle–antiparticle pair (e.g., $W^+W^-$, $b\bar{b}$, $\mu^+\mu^-$,...). These are referred to as the *primary (annihilation/decay) channels*. For each primary channel one can then calculate the spectra of stable SM particles derived after all the intermediate unstable particles are decayed. In any specific DM model the total spectrum is then simply given by a linear sum of contributions from the individual primary channels, weighted by the corresponding branching ratios (BRs), dictated by the DM model,

$$
\frac{dN_{\mathrm{SM}}}{dE} = 
\begin{cases}
\displaystyle\sum_{\mathrm{prim}} \frac{\langle\sigma v\rangle_{\mathrm{prim}}}{\langle\sigma v\rangle_{\mathrm{tot}}} \frac{dN_{\mathrm{SM}}^{\mathrm{prim}}}{dE} \equiv \sum_{\mathrm{prim}} \mathrm{BR}_{\mathrm{prim}} \frac{dN_{\mathrm{SM}}^{\mathrm{prim}}}{dE} & \text{(DM annihilation)}, \\[2ex]
\displaystyle\sum_{\mathrm{prim}} \frac{\Gamma_{\mathrm{prim}}}{\Gamma_{\mathrm{tot}}} \frac{dN_{\mathrm{SM}}^{\mathrm{prim}}}{dE} \equiv \sum_{\mathrm{prim}} \mathrm{BR}_{\mathrm{prim}} \frac{dN_{\mathrm{SM}}^{\mathrm{prim}}}{dE} & \text{(DM decay)}.
\end{cases}
\tag{6.1}
$$

Here $dN_{\mathrm{SM}}^{\mathrm{prim}}/dE$ is the energy spectrum of cosmic rays produced for a single annihilation into a particular primary channel (here, $^{\mathrm{prim}}$ denotes the primary channel and $_{\mathrm{SM}}$ refers to any of the stable cosmic ray SM particles). For instance, $dN_{\gamma}^{W}/dE$ is the spectrum of photons produced in a single DM DM $\to W^+W^-$ annihilation. The branching ratios are explicitly defined in terms of $\langle\sigma v\rangle_{\mathrm{prim}}$ and $\Gamma_{\mathrm{prim}}$, the thermally averaged annihilation cross section and the decay rate for the selected primary channel, while $\langle\sigma v\rangle_{\mathrm{tot}}$ and $\Gamma_{\mathrm{tot}}$ denote the corresponding total quantities. Note that, since DM is cold, the DM annihilation (decay) can be assumed to occur essentially at rest. Each of the two primary particles therefore has the total energy equal to the mass of the DM particle (half of the DM mass).

   In the remainder of this section we work within the above model-independent framework, assuming dominance of two body final states. This does leave aside the possibilities of 3-body or $n$-body annihilations/decays, the annihilations/decays via intermediate states, as well as the annihilations/decays which do not produce a particle-antiparticle pair but rather a combination of different particles in the final

state (e.g., the DM→ $W^{\pm}\tau^{\mp}$ decay, relevant for gravitinos), etc.[6] In some of these cases, the fluxes can be computed by properly combining the model independent building blocks discussed above. In general, however, a dedicated computation has to be performed.

The exhaustive list of primary channels in the model independent approach consists of:

$$e^+e^-, \ \mu^+\mu^-, \ \tau^+\tau^-, \ \nu_e\bar{\nu}_e, \ \nu_\mu\bar{\nu}_\mu, \ \nu_\tau\bar{\nu}_\tau,$$
$$q\bar{q}, \ c\bar{c}, \ b\bar{b}, \ t\bar{t}, \ \gamma\gamma, \ gg, \tag{6.2}$$
$$W^+W^-, \ ZZ, \ hh,$$

where $q = u, d, s$ denotes a light quark[7] and $h$ is the Standard Model Higgs boson. Some channels (such as $\gamma\gamma$, $\nu\bar{\nu}$, $gg$) are 'unusual' as they are often suppressed in many models (in particular, since DM is neutral, its $\gamma\gamma$ yield is generically expected to be suppressed and as a consequence the 3-body and higher-order effects mentioned above can become relevant), however, from a model-independent point of view they are as viable as any other.

The primary particles' decays, showerings and hadronizations are dictated by the SM physics. In most studies the predictions for this part are obtained using one of the Monte Carlo simulation programs designed for collider physics [425–427], appropriately modified to take into account the fact that particles that are considered stable on collider scales (such as muons or neutrons) have enough time to decay in the cosmological environments.[8] `Pythia` is often the default choice, and is also at the base of the results presented here. `Herwig` has also been employed in the literature, while `Geant4` has been used for the specific case of annihilations/decays happening in the dense matter of the Sun, as we will discuss in section 6.9. The algorithms implemented in `Herwig`, `Pythia`, and other codes, differ both in the treatment of parton showers as well as hadronization. The difference in the predictions due to a choice of the numerical tool thus gives us some estimate of theoretical uncertainties. In most cases these are of the order of 20% [425, 426], although larger discrepancies can appear in some specific cases. It is also important to remember that the above tools have been designed and tuned for the energies typical in collider physics (GeV-TeV). Extrapolations to lower or higher energies (i.e., lighter or heavier DM) in general reduce the reliability of the results. Recently, there were thus several dedicated efforts aimed at extending the predictions for spectra in the directions of light and heavy DM [428].

The end product of simulations are the DM produced fluxes of energetic cosmic rays $e^{\pm}, \bar{p}, \bar{d}, \gamma, {}^{(\!-\!)}\nu_{e,\mu,\tau}$. Selected examples of these *primary cosmic ray spectra* are illustrated in fig. 6.3. The spectra vary significantly between different primary channels (different SM particles first produced in the DM annihilation/decay). However, they all exhibit a 'bump-like' shape, marked by a high-energy cutoff at the DM particle mass and a gently diminishing tail at lower energies, potentially accompanied by additional humps.

It is instructive to examine qualitatively a few cases, and reconstruct the origins of different features appearing in the spectra. Let us start with the DM DM $\rightarrow W^+W^-$ annihilations and the induced $e^+$ spectra (dashed blue line in the left panel in fig. 6.3). The $W$'s can decay leptonicaly ($W^{\pm} \rightarrow \ell^{\pm}{}^{(\!-\!)}\nu_\ell$) or hadronicaly ($W^{\pm} \rightarrow q\bar{q}'$). The leptonic $W^{\pm} \rightarrow e^{\pm}{}^{(\!-\!)}\nu_e$ decays directly produce the high-energy electrons or positrons that can, for appropriate kinematical configurations, carry almost all of the energy released in the process. This corresponds to the 'wedge' in the spectrum at $x \sim 1$, visible in the dashed blue line. A similar feature is present for the spectra of neutrinos (not shown in the figure). The leptonic $W^{\pm}$ decays into muons and taus produce, upon further decay of these heavy leptons, more $e^{\pm}$: since in these

---

[6]We will consider some peculiar model-dependent spectra of gamma rays in section 6.2.4.

[7]The light quarks are treated collectively since they lead to very similar spectra, at least for $M \gtrsim$ few GeV.

[8] In Monte Carlo generators the $s$-wave non-relativistic DM DM annihilation is modeled using the equivalent process: the decay of resonance $X$ that has twice the DM mass, $M_X = 2M$, and decays into the appropriate pair of Standard Model particles.

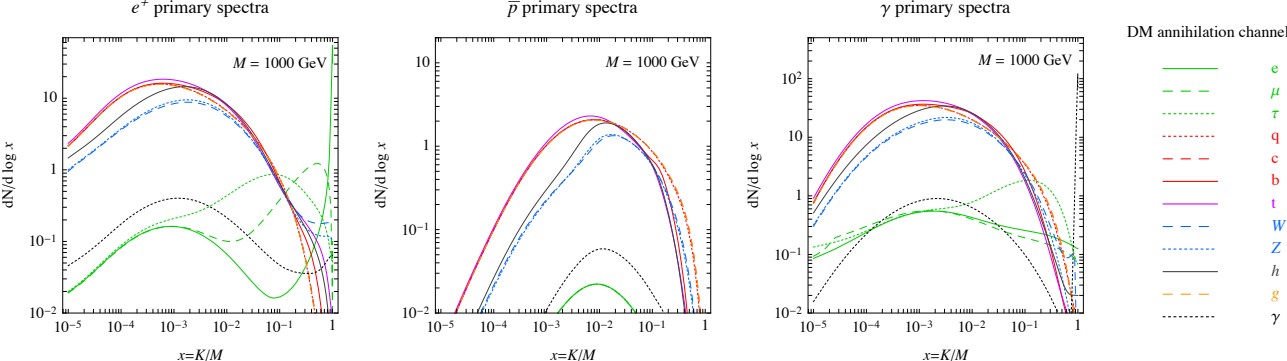

Figure 6.3:  *Examples of **primary cosmic ray spectra** from DM annihilations, for DM with a mass of 1 TeV, and for different primary channels. The spectra are expressed as a function of $x = K/M$, where $K$ is the kinetic energy of the indicated final state particle.*

cases a significant part of the energy is carried away by the neutrinos, the resulting $e^\pm$ will contribute to the $x < 1$ part of the spectrum. The hadronic $W^\pm$ decays produce energetic quarks: they, in turn, decay, fragment and hadronize, producing a host of low energy particles that ultimately decay generating low energy $e^\pm$. They contribute to the prominent hump in the spectra at $x \ll 1$.

In the $\gamma$-ray spectra (right panel in fig. 6.3), one sees another wedge at $x \sim 1$ for the $W$ primary channel. This corresponds to hard photons radiated by the charged $W^\pm$ (this feature is not present for the $Z$ primary channel). The prominent bump at $x \ll 1$ corresponds to all the photons produced in the subsequent cascades, notably the decays of light mesons created during hadronization (e.g., $\pi^0 \to \gamma\gamma$).

As another example, let us consider the case of DM DM $\to e^+ e^-$ annihilations (solid green line in the left panel in fig. 6.3). In the $e^+$ spectrum we can easily recognize the expected $x = 1$ spike, smeared towards lower $x$ by final state radiation. The wide bump around $x \sim 10^{-3}$ might be surprising at first sight: it is due to electroweak emissions (discussed in section 6.1.2 below).

The $\bar{p}$ spectra (middle panel in fig. 6.3) exhibit a more self-similar shape, largely independent from the primary channel. Anti-protons are produced at the end of the hadronization process, which 'washes away' the peculiarities related to the initial particles, where only the purely leptonic primary channels are distinct from the rest. The small production of $\bar{p}$ from the purely leptonic $e^+ e^-$ and $\mu + \mu^-$ channels is due to electroweak emissions, while $\tau^+ \tau^-$ also has hadronic decay modes.

Similar considerations allow to understand many of the qualitative features of the other spectra as well.

## 6.1.1   Production of light nuclei

Antideuterons (and antihelium nuclei) deserve a separate discussion [429]. Antideuterons can be produced via the fusion of a $\bar{p}\bar{n}$ pair, a process usually described by a simple 'coalescence model'. The idea behind this approach is very simple: the anti-nucleons produced in the DM annihilation/decay process can merge to form an anti-nucleus, if the difference in their momenta is less than a certain threshold, the effective quantity called the coalescence momentum $p_{\rm coal}$. The value of $p_{\rm coal}$ is a free parameter of the model that has to be determined from a comparison with the experimental data, for instance from the $e^+ e^- \to \bar{d}X$ collider reactions (here, $\bar{d}$ denotes an anti-deuteron, and $X$ inclusively any other final state). The naïve estimate $p_{\rm coal} \approx \sqrt{m_N B_d} \approx 46$ MeV, with $m_N$ the nucleon mass and $B_d = 2.2$ MeV the deuteron binding energy, is not too far from the typically adopted values for $p_{\rm coal}$, which range from 66 to 226 MeV.

One way of implementing the above prescription is to take the Monte-Carlo-produced spectra of $\bar{p}$ and $\bar{n}$ discussed above (they are typically equal, barring unusual isospin violation) and combine them

using the coalescence prescription. One can show that the resulting $\bar{d}$ spectrum then reads

$$\frac{dN_{\bar{d}}}{dT_{\bar{d}}} = \frac{4\pi}{3} p_{\text{coal}}^3 \frac{m_{\bar{d}}}{m_{\bar{n}} m_{\bar{p}}} \frac{1}{\sqrt{T_{\bar{d}}^2 + 2m_{\bar{d}} T_{\bar{d}}}} \left(\frac{dN_{\bar{p}}}{dT_{\bar{p}}}\right)_{T_{\bar{p}}=T_{\bar{d}}/2} \left(\frac{dN_{\bar{n}}}{dT_{\bar{n}}}\right)_{T_{\bar{n}}=T_{\bar{d}}/2}, \tag{6.3}$$

where $T$ denotes the kinetic energy. This formula shows that the $\bar{d}$ yield depends significantly on the value of $p_{\text{coal}}$. The $\bar{d}$ spectrum is also 'doubly suppressed' with respect to the already small $\bar{p}$ and $\bar{n}$ spectra: as a rule of thumb, the $\bar{d}$ yield is found to be a factor of $\mathcal{O}(10^{-5})$ smaller than the $\bar{p}$ one.

The above procedure assumes, however, that the events are spherically symmetric, i.e., an isotropic emission of the anti-nucleons from the DM annihilation/decay process. This assumption is violated in processes in which the anti-nucleons are produced from boosted initial particles. For instance, for DM DM $\rightarrow W^+W^-$ and for a large DM mass, the anti-nucleons are produced in the boosted back-to-back jets produced by the $W^\pm$, enhancing the probability of having $\bar{p}\bar{n}$ pairs with small momentum difference. In order to have the correct prediction for the $\bar{d}$ yields, the details of the angular distribution of the anti-nucleons in the final state, together with possible (anti-)correlations between them, must be taken into account. This has to be done by computing the $\bar{d}$ spectrum numerically, enforcing the coalescence prescription on an event-by-event basis in the Monte Carlo itself. This feature is included in `Pythia` since v8.240.

Identical considerations apply to the anti-helium production [430], in its two possible isotopic forms, $^3\overline{\text{He}}$ and $^4\overline{\text{He}}$. Each additional coalescing anti-nucleon brings an additional factor of $dN/dT$, while in the naive formula in eq. (6.3) the $p_{\text{coal}}^3$ factor is replaced by $p_{\text{coal}}^{3(A-1)}$, where $A$ is the mass number. The final yield can be expected to be suppressed by an additional factor of $\mathcal{O}(10^{-5})$ per added anti-nucleon: in practice, only $^3\overline{\text{He}}$ may be produced in non-negligible amounts. Its coalescence can proceed from either a $\bar{p}\bar{p}\bar{n}$ or a $\bar{p}\bar{n}\bar{n}$ triplet. In the former case the $^3\overline{\text{He}}$ is formed directly, but the efficiency is suppressed by Coulombian repulsion between the two anti-protons. In the latter case, $^3\overline{\text{He}}$ is the result of the formation of an anti-tritium that subsequently decays into a $\overline{\text{He}}$ in a process that, given the typical propagation scales, can be considered to be instantaneous. In the simulations, the coalescence prescription needs to be generalized appropriately: one can require that all the three differences among the anti-nucleon momenta in the rest frame of the center of mass are smaller than $p_{\text{coal}}$, or that the three momenta lie inside a minimum bounding sphere in momentum space with a diameter determined by $p_{\text{coal}}$. In practice, the differences in the yields obtained using different prescriptions are small, compared to other uncertainties.

The value of $p_{\text{coal}}$ is hard to establish, given the scarcity of the $\overline{\text{He}}$ collider data. One option is to determine it from the $p_{\text{coal}}$ value for $\bar{d}$ by scaling it with the binding energy as $p_{\text{coal}}^{\text{He}} \approx \sqrt{B_{\text{He}}/B_d}\, p_{\text{coal}}^d$. The values typically adopted in the literature therefore span a large range $p_{\text{coal}}^{\text{He}} \approx 200 - 400\,\text{MeV}$. More sophisticated methods than the coalescence prescription have also been put forward and employed in the recent literature.

## 6.1.2 Electroweak emissions

Among the different processes occurring after the production of primary particles, the electroweak emissions are particularly relevant. This term denotes the radiation of $W$ or $Z$ electroweak gauge bosons from outgoing particles. (The $W$ and $Z$, of course, subsequently decay.) This emission is a higher order process with respect to the 2-body annihilation/decay and is therefore suppressed by an additional weak coupling. However, such bremsstrahlung corrections are enhanced by one or more powers of $\ln(M/M_W)$ logarithms, which become large for $M \gg M_W$, compensating the suppression. Consequently, the electroweak emissions have only recently been identified as significant for DM indirect detection [431], particularly as interest in TeV DM masses has grown. For instance, these effects were absent in `Pythia` versions prior to v8.176 (2013). This prompted the authors of [425] to add them, working at single emission order, including vertex corrections, by 'post-processing' the `Pythia` output. Subsequent `Pythia` versions have

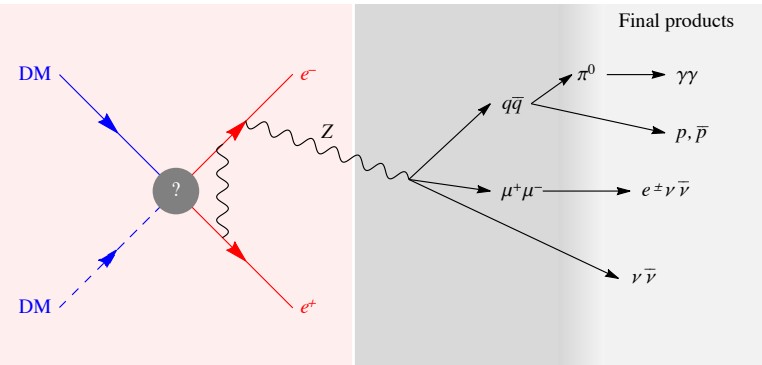

Figure 6.4: ***Illustration of electroweak corrections***, *for the case of a primary annihilation into $e^+e^-$. In the absence of any electroweak emissions, the resulting primary cosmic ray spectrum would consist of just the $e^+e^-$ pair, i.e., a delta function at $x = K/M = 1$ in the $e^\pm$ spectrum. The emission of a weak boson (a Z in this case), as well as the accompanying virtual boson exchange (vertex correction), modify the spectrum. The Z decays, which then gives lower energy $e^\pm$, as well as other particle species. The hadronic decays of the gauge boson produce quarks that hadronize into, among other things, anti-protons, which can therefore emerge from a purely leptonic DM channel.*

gradually incorporated these effects, including higher order corrections. Furthermore, these effects have been explicitly addressed in more recent codes [427, 428], including those developed to study extremely heavy DM.

Phenomenologically, the effects of the electroweak radiation can be particularly relevant for the leptonic and $\gamma\gamma$ primary channels. The emissions of $W$'s and $Z$'s, and their subsequent decays, lead to additional particles in the final state, and can therefore modify significantly the flux of $\gamma$'s and $e^\pm$ at lower energies, $E \ll M$. Moreover, the $W/Z$ radiation leads to a $\bar{p}$ contribution, which would have been absent in the leptonic and $\gamma\gamma$ primary channels, were the electroweak corrections to be neglected. This also holds true for the neutrino primary channels, where the electroweak emissions result in nonzero $e^\pm$'s, $\gamma$, and $\bar{p}$ primary ray spectra.

There are other instances where the electroweak corrections are important. For instance, the annihilation rates of Majorana DM particles into light fermions are helicity suppressed. Inclusion of the electroweak emissions lifts the suppression, and allows the annihilation to proceed in an $s$-wave, implying a much larger cross section and modified spectra. It is also good to keep in mind that the EW emission can occur from the final state particles, from the internal states running in the loop, or even from the initial DM states, if DM is part of an EW multiplet (its neutral component) [431].

## 6.2 Gamma rays

In this section, we examine the calculation of observables related to prompt gamma rays produced in DM annihilations/decays. We defer the discussion of secondary radiation, which can also have energy in the domain of gamma rays, to section 6.7.

To search for photon signals from DM there are a number of varied *"best targets"*. Ideally, these are regions with high DM densities or regions with low astrophysical 'backgrounds', resulting in a favorable signal-to-noise ratio. The demarcation between the two requirements is not clear cut: environmental factors may render certain regions more suitable than others, despite comparable DM densities and backgrounds.

Typically, the experimental focus is on:

- ○ **The Milky Way Galactic Center (GC)**, or small regions around or just outside it, often referred to as the Galactic Center Halo (GCH).

- ○ **Wide regions of the Galactic Halo (GH)**, up to several tens of degrees wide in latitude and longitude, and away from the Galactic plane. The average DM density is lower in these regions than it is close to the GC. However, the astrophysical contribution is also much lower, since this is heavily concentrated in the Galactic plane. The GH regions are also promising for the detection of the secondary radiation, given that the DM-produced CRs have less competition from the CRs of astrophysical origin than they do in the Galactic plane.

- ○ **Globular clusters (GloCs)**, i.e., dense agglomerates of stars with the total mass of order $10^5 \, M_\odot$, embedded in the Milky Way galactic halo or in the subhalo of dwarf galaxies.[9] The DM content (and its distribution) in GloCs is still subject to debate [432]. The interest in GloCs stems from four key aspects: i) they could have originated from a primordial DM sub-halo, possibly retaining some trapped DM; ii) the density of baryonic matter could generate a DM concentration via attraction, enhancing DM annihilation flux, and even have an effect on the DM halo of the host galaxy; iii) they are closer than dwarf galaxies and would therefore lead to a brighter DM annihilation flux on Earth; iv) they might host an Intermediate Mass Black Hole (IMBH), which could also significantly enhance the DM annihilation flux.

- ○ **Subhalos of the GH.** The positions of subhalos are, however, unknown a priori. See the discussion in section 2.4.

- ○ **Satellite galaxies of the Milky Way,** often of the dwarf spheroidal (dSph) class, such as Sagittarius, Segue 1, Draco, and several others. These satellite galaxies are star deprived and are believed to be DM dominated. See the discussion in section 2.2.3.

- ○ **Large scale structures in the relatively nearby Universe,** such as the galaxy clusters: the Virgo, Coma, Fornax, and Perseus clusters, as well as several others. In such larger objects the DM particles have higher velocities on average, so that these are good probes of DM with velocity-suppressed $p$- or $d-$wave annihilations [433].

- ○ **The entire Universe,** by measuring the *cosmological* flux of redshifted $\gamma$-rays at the position of the Earth, and originating from DM annihilations in all the halos along the line of sight, spanning the complete recent history of the Universe. This flux is often termed as 'extragalactic', though one needs to be careful not to confuse it with the flux from specific large structures located outside our galaxy, and mentioned in the previous item. Alternatively, it is described as 'isotropic' (IGRB, Isotropic Gamma-Ray background), though it is only nearly isotropic (see section 6.2.3).

In the rest of this section, we will discuss the main quantities related to gamma-ray observations from these various targets.

## 6.2.1   $\gamma$ from the Milky Way

The *differential flux of photons* from a given angular direction $d\Omega$, produced by the annihilations of self-conjugated DM particles (e.g. Majorana fermions) inside the Milky Way, is given by

$$\frac{d\Phi_\gamma}{d\Omega \, dE} = \frac{1}{2} \frac{r_\odot}{4\pi} \left( \frac{\rho_\odot}{M} \right)^2 J \langle \sigma v \rangle \frac{dN_\gamma}{dE}, \qquad J = \int_{\text{l.o.s.}} \frac{ds}{r_\odot} \left( \frac{\rho(r(s,\theta))}{\rho_\odot} \right)^2 \qquad \text{(DM annihilation)}, \qquad (6.4)$$

---

[9]The differentiation between a (large) GloC and a (small) dwarf galaxy is nuanced; typically, the former exhibits greater density, an abundance of stars, and proximity to Earth.

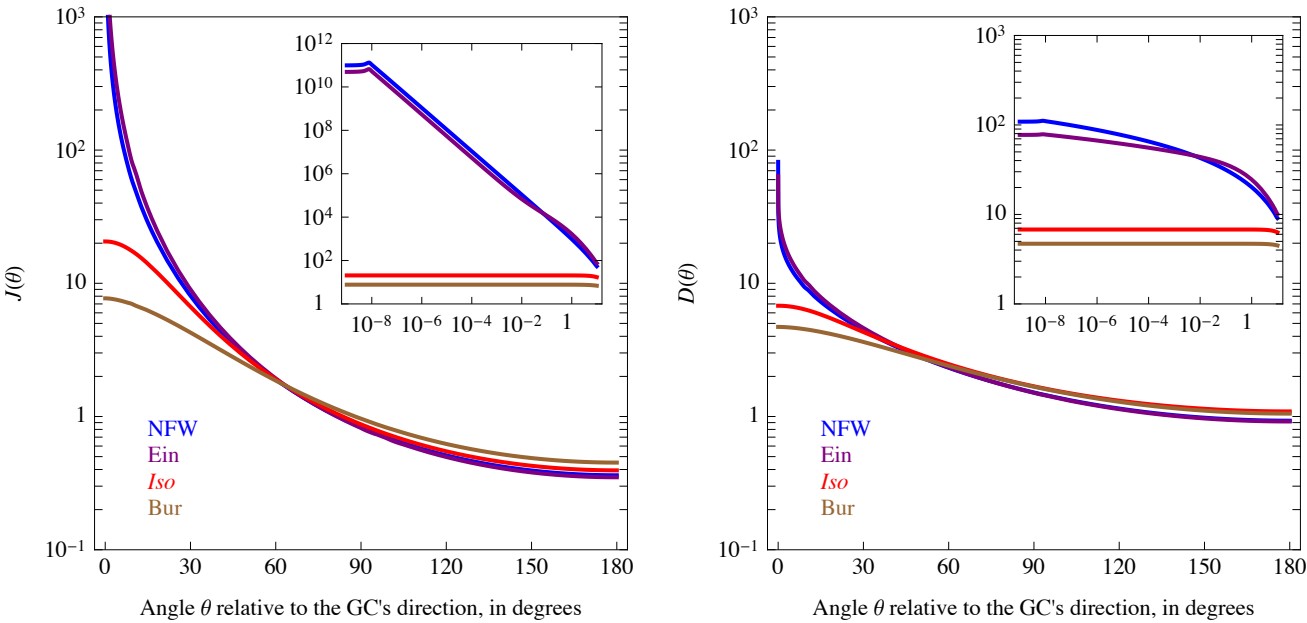

Figure 6.5:  $J(\theta)$ *for annihilating (left) and* $D(\theta)$ *for decaying (right)* **Dark Matter**, *for different DM profiles introduced in section 2.2, with the color coding given in the legend (from bottom to top in the inset: Burkert, Isothermal, Einasto, NFW).*

while for a decaying DM we have, similarly,

$$\frac{d\Phi_\gamma}{d\Omega\,dE} = \frac{r_\odot}{4\pi}\frac{\rho_\odot}{M}D\,\Gamma\frac{dN_\gamma}{dE}, \qquad D = \int_{\text{l.o.s.}} \frac{ds}{r_\odot}\frac{\rho(r(s,\theta))}{\rho_\odot} \qquad \text{(DM decay)}. \tag{6.5}$$

The radial coordinate $r$, measured from the GC, is given by $r(s,\theta) = (r_\odot^2 + s^2 - 2\,r_\odot\,s\cos\theta)^{1/2}$, while $\theta$ is the aperture angle between the direction of the line of sight (l.o.s., parametrized by variable $s$) and the axis linking the Earth to the GC. The total energy spectrum of photons produced by DM annihilations/decays, denoted as $dN_\gamma/dE$, is computed in eq. (6.1). If DM is not composed of self-conjugate particles (e.g., if DM particle is a Dirac fermion), then $\sigma v$ in eq. (6.4) must be averaged over DM particles and antiparticles. In practice this means that the expression in eq. (6.4) has to be divided by an additional factor of 2, if only particle-antiparticle annihilations are present. It is also important to note that the above formalism, especially the simple definition of the $J$ factor determined solely by the DM density profile, holds true only when the annihilation cross section is position independent. If this is not the case, the annihilation cross section must be included as part of the integrand inside the l.o.s. integral. In particular, if the annihilation cross section is velocity-dependent, which, for instance, happens for $p$-wave or Sommerfeld enhanced annihilations, the calculation of the total flux must include the full DM velocity distribution (see e.g. Boucher et al. (2022) [434]).

The **$J$ factor** in eq. (6.4) and the **$D$ factor** in eq. (6.5) include appropriate factors of $r_\odot$ and $\rho_\odot$ (the Sun–GC distance and the DM density at the location of the Sun, see section 2.2.2) to make them dimensionless.[10] We work in the limit of a spherically symmetric DM halo, so that $J(\theta)$ and $D(\theta)$ are invariant under rotations around the Sun–GC axis, and therefore depend only on the aperture angle $\theta$. Fig. 6.5 shows the values of $J$ and $D$ factors as functions of $\theta$ for a number of plausible galactic DM profiles, which were discussed in section 2.2. Since the $J$ factor depends quadratically on the DM density

---

[10]Alternatively, sometimes two analogous factors are defined, the DM annihilation factor $\mathcal{J} = \int_{\text{l.o.s.}} \rho^2(r) = r_\odot\rho_\odot^2\,J$, having units of GeV$^2$/cm$^5$, and the DM decay factor $\mathcal{D} = \int_{\text{l.o.s.}} \rho(r) = r_\odot\rho_\odot\,D$, having units of GeV/cm$^2$.

(compared to the linear dependence in the $D$ factor), this translates to a larger sensitivity to the DM density profile.

Eq.s (6.4) and (6.5) are useful if one needs the flux of gamma rays from a given direction. More often often than not, one needs instead the flux integrated over a solid angle $\Delta\Omega$, for example, the window of observation or the resolution of the telescope. The $J$ factor is then replaced by the *integrated $J$ factor* for such a region, defined simply as $J(\Delta\Omega) = \int_{\Delta\Omega} J \, d\Omega$. One can also define the *average $J$ factor* as $\bar{J}(\Delta\Omega) = \left(\int_{\Delta\Omega} J \, d\Omega\right)/\Delta\Omega$. Analogous expressions hold also for the integrated and average $D$ factors. The following simple formulæ hold for representative solid angle regions, symmetric around the GC: a disk of aperture $\theta_{\max}$ centered around the GC; an annulus centered around the GC, $\theta_{\min} < \theta < \theta_{\max}$; and a generic region defined in terms of the galactic latitude $b$ and longitude $\ell$,[11]

$$\Delta\Omega = 2\pi \int_0^{\theta_{\max}} d\theta \, \sin\theta, \qquad \bar{J} = \frac{2\pi}{\Delta\Omega} \int d\theta \, \sin\theta \, J(\theta), \qquad \text{(disk)}$$

$$\Delta\Omega = 2\pi \int_{\theta_{\min}}^{\theta_{\max}} d\theta \, \sin\theta, \qquad \bar{J} = \frac{2\pi}{\Delta\Omega} \int d\theta \, \sin\theta \, J(\theta), \qquad \text{(annulus)} \qquad (6.7)$$

$$\Delta\Omega = 4 \int_{b_{\min}}^{b_{\max}} \int_{\ell_{\min}}^{\ell_{\max}} db \, d\ell \, \cos b, \quad \bar{J} = \frac{4}{\Delta\Omega} \iint db \, d\ell \, \cos b \, J(\theta(b,\ell)), \quad (b \times \ell \text{ region}).$$

Completely analogous formulæ hold for the case of DM decay.

## 6.2.2   $\gamma$ from dwarf galaxies

The same formalism applies to dwarf satellite galaxies [58] (discussed in section 2.2.3). One defines the integrated $\mathcal{J}$ and $\mathcal{D}$ factors, omitting the factors of $r_\odot$ and $\rho_\odot$ used for the MW case in eq.s (6.4) and (6.5). These encode the $\gamma$ emissions produced along the line of sight, pointing from Earth towards the dwarf galaxy, and within a small disk (of aperture $\theta^\star$) that contains the source. The $\gamma$ flux at the position of the Earth is thus given by

$$\frac{d\Phi_\gamma}{dE_\gamma} = \frac{dN_\gamma}{dE_\gamma} \frac{1}{4\pi} \begin{cases} \mathcal{J}_{\text{dSph}} \langle\sigma v\rangle/2M^2 & \mathcal{J}_{\text{dSph}} = 2\pi \int_0^{\theta^\star} d\theta \, \sin\theta \int_{\text{l.o.s.}} ds \, \rho^2(s) & \text{(DM annihilation)}, \\ \mathcal{D}_{\text{dSph}} \, \Gamma/M & \mathcal{D}_{\text{dSph}} = 2\pi \int_0^{\theta^\star} d\theta \, \sin\theta \int_{\text{l.o.s.}} ds \, \rho(s) & \text{(DM decay)}. \end{cases} \quad (6.8)$$

As before, the photon flux depends quadratically (linearly) on the DM profile in the dwarf galaxy, $\rho$, for DM annihilations (decays). The DM profile is determined via stellar kinematical data. Since stars are scarce in these DM dominated systems, the determination of the DM profile is one of the most critical points and the dominant source of uncertainty. In some extreme cases, e.g. the Segue I or Leo IV ultrafaint dwarfs, discarding a single star in the sample of tracers associated with a given galaxy can modify significantly its reconstructed DM content and DM density profile, and change the extracted $\mathcal{J}_{\text{dSph}}$ and $\mathcal{D}_{\text{dSph}}$ even by orders of magnitude. In many other cases, however, the impact is much more limited, essentially since the whole source is included in the observed disk.

Another subtle point is the choice of the maximum angle of integration $\theta^\star$, i.e., the aperture of the cone inside which one assumes to be collecting the DM signal from the dwarf. Several choices have been discussed and adopted in the literature: (i) $\theta^\star = \theta_{\max}$, where $\theta_{\max}$ is the position of the outermost star that can be associated with the dwarf; (ii) $\theta^\star = \theta_{\text{HL}}$, the angle corresponding to the half-light radius as

---

[11]The galactic polar coordinates $(d, \ell, b)$ are defined as

$$x = d\cos\ell\cos b, \qquad y = d\sin\ell\cos b, \qquad z = d\sin b \qquad (6.6)$$

where the Earth is located at $\boldsymbol{x} = 0$ (such that $d$ is the distance from the Earth); the GC is located at $x = r_\odot$, $y = z = 0$; and the Galactic plane corresponds to $z \approx 0$. Consequently, $\cos\theta = x/d = \cos b \cdot \cos\ell$.

defined in section 2.2.3; (iii) $\theta^\star = \theta_c$, where the critical angle $\theta_c$ is defined in terms of the half-light radius of the dwarf $r_{HL}$ and its distance $d$ as $\theta_c = 2\,r_{HL}/d$ (for annihilation) and $\theta_c = r_{HL}/d$ (for decay), which is a definition that has been shown to minimize the systematic uncertainties, e.g., the ones related to the non-sphericity of the halo; (iv) $\theta^\star$ corresponding to a fixed universal value, irrespective of the size and the distance of the dwarf, with a typical choice $\theta^\star = 0.5°$. Table 2.3 lists determinations of $\mathcal{J}_{dSph}$ and $\mathcal{D}_{dSph}$ for most of the Milky Way dwarf satellites. In the table we report the values for $\theta^\star = 0.5°$, using the most recent determinations available in the literature, and refer the reader to dedicated studies for more details [58].

Note that, since the line of sight to satellite galaxies passes through the DM halo of the Milky Way, the latter will also contribute to the total integral. Depending on the position of the satellite galaxy and on the assumptions on the galactic DM profile, this contribution can be negligible or dominant (see, e.g., Bonnivard et al. (2015) [58] for some examples).

### 6.2.3   Cosmological $\gamma$

In the case of the *cosmological $\gamma$-ray flux* [435], the calculation of the flux perceived on Earth grows more complex for two main reasons: the need to account for 'cosmological dimming' caused by the expansion of the universe, and the fact that, in contrast to the galactic environment, the absorption of gamma rays cannot be overlooked over cosmologically large distances.

The differential flux of cosmological gamma rays at redshift $z$ is given by

$$\frac{d\Phi_{cosmo\gamma}}{dE_\gamma} = \frac{1}{E_\gamma} \int\limits_z^\infty dz' \frac{j_{cosmo\gamma}(E'_\gamma, z')}{H(z')(1+z')} \left(\frac{1+z}{1+z'}\right)^3 e^{-\tau(E_\gamma, z, z')}, \tag{6.9}$$

where $E'_\gamma = E_\gamma(1+z')/(1+z)$ is the energy of a photon at redshift $z'$, such that it has energy $E_\gamma$ at redshift $z$. The denominator $H(z')(1+z')$ converts the redshift interval into a proper distance interval (the integral can be thought of as integrating over time all the past photon emissions, and then converting it into redshift). Here, $H(z)$ is the Hubble rate, see appendix C. The factor $[(1+z)/(1+z')]^3$ accounts for the cosmological dimming of intensities due to the dilution of the number density of emitted photons.

The function $\tau(E_\gamma, z, z')$ is the optical depth encountered by a $\gamma$-ray between the emission redshift $z'$ and the collection redshift $z$, resulting in the absorption factor $e^{-\tau}$. The absorption occurs due to several kinds of interactions between the $\gamma$-ray and the diffuse intergalactic gas and light. Roughly ranked by importance as the $\gamma$-ray energy increases, these interactions include: photoionization, Compton scattering and pair production on the intergalactic gas; photon-photon scattering and photon-photon pair production on the extragalactic background light (EBL), which consists of UV, optical and infrared light emitted by stars[12], and the far-infrared light emitted by dust particles reprocessing stellar light, as well as photon-photon interactions on the CMB. We do not review all these processes here but instead refer the reader to the relevant extensive literature [425, 436]. As a rule of thumb, the universe is mostly transparent for MeV photons, even if they are emitted at very high redshifts. Photons with keV and GeV energy are instead totally absorbed if emitted at $z \gtrsim 100$, and partially absorbed for lower $z$. TeV photons are absorbed if emitted at $z \gtrsim 0.1 - 10$, depending on the EBL details, and 10 TeV photons are totally absorbed unless emitted locally.

Eq. (6.9) is useful if one is interested in the impact of gamma-rays at the redshift $z$, e.g. for calculating the energy deposition in the intergalactic medium from all the annihilations or decays that occurred at

---

[12]Scattering on solar light is also relevant and introduces a local anisotropy, see S. Balaji (2022) in [436], although the effect is very small.

earlier times. For gamma rays observations at the present time, i.e. $z = 0$, eq. (6.9) simplifies to

$$\frac{d\Phi_{\text{cosmo}\gamma}}{dE_\gamma} = \frac{1}{E_\gamma} \int\limits_0^\infty dz' \frac{j_{\text{cosmo}\gamma}(E'_\gamma, z')}{H(z')(1+z')^4} \, e^{-\tau(E_\gamma, z')}. \tag{6.10}$$

The function $j_{\text{cosmo}\gamma}$ encodes gamma-ray emissivity for annihilating or decaying DM:

$$j_{\text{cosmo}\gamma}(E'_\gamma, z') = E'_\gamma \frac{dN_\gamma}{dE'_\gamma} \begin{cases} B(z')\langle\sigma v\rangle[\bar\rho_{\text{DM}}(z')/M]^2/2 & \text{(DM annihilation)}, \\[2mm] \Gamma \, \bar\rho_{\text{DM}}(z')/M & \text{(DM decay)}, \end{cases} \tag{6.11}$$

where $dN_\gamma/dE'_\gamma$ is the spectrum of the prompt photons in eq. (6.1), and $\bar\rho_{\text{DM}}(z) = (1.26 \text{ keV/cm}^3)(1 + z)^3$ is the average cosmological DM density, see eq. (1.1). The 'boost factor' $B(z)$ accounts for the enhancement in the rate of DM annihilations due to DM clustering compared to a homogeneous universe: this will be further discussed in section 6.8.1 below.

### Auto-correlations and cross-correlations of the cosmological $\gamma$ flux

In a first approximation the cosmological $\gamma$-ray flux from DM can be considered isotropic. However, it also has a subdominant intrinsic anisotropic component, since it is produced by large-scale DM structures that are distributed anisotropicaly throughout the Universe (see section 1.3.1). This characteristic can be used as a tool to search for DM and, if no signal is detected, to impose constraints [437].

The underlying idea is to search for spatial correlations between two emissions related to DM. Beyond the cosmological $\gamma$-ray flux, DM can also emit lower-energy photons, such as $X$-rays or radio-waves, through secondary processes like inverse Compton scattering or synchrotron radiation (see section 6.7). Additionally, DM can be traced through its gravitational effects. These include the gravitational weak-lensing signal (cosmic shear, see section 1.2.1) produced by DM large structures, the clustering in the distribution of galaxies (which correlates with the presence of DM), and the 21 cm emission of hydrogen gas clouds, whose positions also trace the presence of DM.

One can therefore search for: (i) **Autocorrelations of electromagnetic emissions from DM:** This includes autocorrelations in the cosmological $\gamma$-ray flux or radio flux, and is akin to analyses performed on the CMB (see section 1.3.3). Detecting a non-zero autocorrelation due to DM would shed light on the distribution of DM on very large scales. By comparing the observed signal with the signal predicted using realistic DM distributions, one could impose significant constraints on DM properties, such as the annihilation cross-section or the decay rate (see, e.g., Fornengo et al. (2011) [437]). (ii) **Cross-correlations between two electromagnetic signals:** These correlations include radio-$\gamma$, radio-$X$ and $X$-$\gamma$ cross-correlations. (iii) **Cross-correlations between electromagnetic emissions and gravitational tracers:** Examples include $\gamma$-shear, $\gamma$-galaxy or $\gamma$-21cm cross-correlations. Strategy (iii) is particularly interesting, because it would relate an unambiguous manifestation of DM being an elementary particle (its electromagnetic emission) with a direct probe of the DM's existence in the Universe through its gravitational effects.

## 6.2.4 Sharp spectral features in prompt $\gamma$-rays

The annihilations (or decays) of DM particles in specific channels or in specific models can generate distinct, sharp features in the spectra of the produced gamma-rays [423]. These features are potentially useful for detection because they can be easily distinguished from the predicted astrophysical background, especially since, unlike charged cosmic rays, the features in the gamma-ray spectra are not deformed by astrophysical propagation. In this subsection, we focus on DM annihilation for the sake of simplicity, but

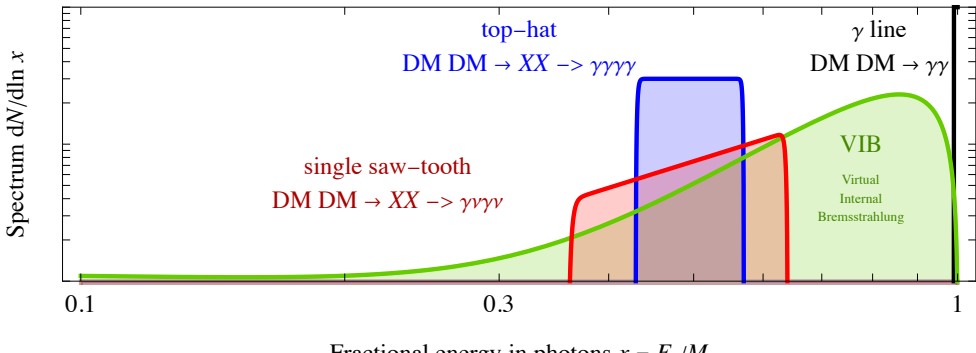

Figure 6.6:    *Illustration of notable γ-ray **spectral features** possibly produced by DM annihilation or decays.*

a completely analogous discussion would apply to the case of DM decay, with corresponding adjustments to the γ-ray energies.

The simplest of these features is a **gamma-ray line**, i.e., a monochromatic spectrum with a fixed energy, depicted by the black line in fig. 6.6. Such a line can be produced, for example, by the process DM DM → γγ. Since the DM particles are almost at rest in the galactic and cosmological environments (being cold), they possess negligible kinetic energy, and thus the emitted line is at the energy

$$E_\gamma = M. \tag{6.12}$$

For heavy DM the electroweak emissions (see section 6.1.2) smear the peak of the 'delta function', and, more importantly, introduce an additional continuum at $E_\gamma \ll M$. These effects are visible in the example spectrum in fig. 6.3 (dotted black line). As already mentioned above, DM annihilations into photons occur only via some subdominant mechanism, typically involving charged particle loops, resulting in a smaller cross section (often reduced by a loop factor) compared to direct annihilation. This renders the detection more challenging. Nevertheless, the spectral feature is so pronounced that the monochromatic γ-ray line has been regarded for a long time as the 'smoking gun' for DM detection. For a real-world example, see section 8.2.7.

In most models, the loop of charged particles mediating the DM DM → γγ process allows to replace one of the γ with another neutral SM particle in the final state, e.g., a $Z$ or a Higgs boson. Due to the finite masses of $Z$ and $h$, this leads to additional γ-lines at lower energies,

$$E_\gamma = M \left( 1 - \frac{M_{Z,h}^2}{4M} \right). \tag{6.13}$$

Another distinct feature in γ rays is produced by **Virtual Internal Bremsstrahlung (VIB)**, which arises when a photon is emitted from a virtual charged particle that mediates DM annihilation. By bringing the charged particle's propagator closer to being on-shell, VIB favors the emission of higher-energy photons around the maximal photon energy $E_\gamma \lesssim M$, particularly when the charged particle's mass is not very different from the DM mass. Consequently, the VIB spectrum exhibits a triangular shape: rising up to a sharp cut-off at an energy corresponding to $M$, as illustrated by the green curve in fig. 6.6. In addition, VIB turns the annihilation into a three-body process, thereby eliminating the helicity suppression when present. This can be phenomenologically very important in certain cases, with the VIB cross section much higher than the cross section for the tree-level 2-body annihilation.

A DM that annihilates into two metastable bosons $X$ with mass $m_X$, which subsequently decay into photon pairs (DM DM $\rightarrow XX \rightarrow \gamma\gamma\gamma\gamma$), produces a **'top-hat'** or **'box-shaped'** spectrum. The upper and lower photon energies $E_\pm$ and the width $\Delta E_\gamma$ of this spectrum are given by

$$E_\pm = \frac{M}{2}\left(1 \pm \sqrt{1 - \frac{m_X^2}{M^2}}\right), \qquad \text{and} \qquad \Delta E_\gamma = \sqrt{M^2 - m_X^2}. \tag{6.14}$$

This is illustrated by the blue curve in fig. 6.6. In the limit where $m_X \lesssim M$, the phase space becomes restricted, and the spectrum narrows down to a thin line.

A DM that annihilates according to the process DM DM $\rightarrow XX \rightarrow \gamma\nu\,\gamma\nu$, i.e., into two metastable Dirac fermions $X$, which subsequently decay into a photon and another fermionic state (such as a neutrino), produces a **'single saw-tooth'** spectrum. This spectrum shares similarities with the top-hat spectrum, but features a slanted top (see the red curve in fig. 6.6). The slope of the slanted top is determined by the spin polarization of $X$. The masses must satisfy $M > m_X > m_\nu \approx 0$. The formulæ for the edge energies and the width are the same as those given in eq. (6.14).

## 6.3 Neutrinos

Neutrinos produced from DM annihilations or decays in either galactic or cosmological environments[13] propagate along nearly straight paths with velocity $v \simeq c$, akin to photons. Consequently, the discussion of such neutrino signals follows closely the discussion of gamma ray signals presented in the previous section.

The main targets for observation are the Galactic Center, the Galactic Halo, dwarf galaxies, clusters, and the cosmological signal. However, not all of these targets are easily accessible due to the significantly more challenging detection of neutrinos compared to photons (see section 6.13 for the current status of the searches). The expected neutrino flux from the Milky Way can be expressed using equations analogous to eq.s (6.4) and (6.5), with the same $J$ and $D$ factors. Similarly, the $\mathcal{J}$ and $\mathcal{D}$ factors for dwarf galaxies can be described using eq. (6.8).

A notable difference between photons and neutrinos is that the neutrinos undergo flavor oscillations, causing the fluxes at detection to differ from those at production. Given the observed neutrino masses $m_i$ and mixings $V_{\ell i}$, vacuum oscillations over astrophysical distances $L$ can often be well approximated by the limit $\Delta m^2 L/E \gg 1$, where squared oscillation amplitudes are reduced to classical flavour transition probabilities [438]

$$P_{\ell\ell'} = \sum_{i=1}^{3} |V_{\ell i} V_{\ell' i}|^2 \approx \begin{pmatrix} 0.6 & 0.2 & 0.2 \\ 0.2 & 0.4 & 0.4 \\ 0.2 & 0.4 & 0.4 \end{pmatrix}. \tag{6.15}$$

This means that, for instance, annihilations or decays into $\nu_e$ produce a flavor composition at detection $(\nu_e : \nu_\mu : \nu_\tau) \sim (0.6 : 0.2 : 0.2)$, while the annihilations/decays into either $\nu_\mu$ or $\nu_\tau$ produce $(\nu_e : \nu_\mu : \nu_\tau) \sim (0.2 : 0.4 : 0.4)$. Often, for the sake of simplicity, the assumption is made of democratic composition at detection $(\nu_e : \nu_\mu : \nu_\tau) \sim (1 : 1 : 1)$. This flavour composition is preserved by oscillations and is also motivated by the fact that a composition close to $(1 : 1 : 1)$ arises, after oscillations, if neutrinos originate from pion decays in the source, since $\pi^\pm$ decays equally into $\overset{(-)}{\nu}_e$ and $\overset{(-)}{\nu}_\mu$.

Another hypothetical difference with respect to photons is that neutrinos could interact with DM during their galactic or extragalactic propagation. This is addressed in the next subsection.

---

[13] A distinct class of indirect DM searches with neutrinos focuses on the annihilations or decays in the center of the Sun or the Earth. The phenomenology of these searches is entirely different and is covered in detail in section 6.9.

### 6.3.1   Neutrino/DM interactions?

If DM couples to the SM neutrinos, it can leave a discernible imprint in the diffuse flux of high energy neutrinos detected on Earth [439]. As extra-galactic neutrinos traverse the galactic plane, they may scatter on the DM in the halo, leading to an attenuation of the neutrino flux. The most striking effect is obtained if the scattering is through an $s$-channel resonance, $X$,

$$\nu + \mathrm{DM} \to X^* \to \nu + \mathrm{DM}. \tag{6.16}$$

The resonance $X$ with mass $m_X$ is produced on-shell if the neutrino has the energy

$$E_\nu^* = \frac{m_X^2 - M^2}{2M} = 5\,10^3 \,\mathrm{TeV}\, \frac{m_X^2 - M^2}{\left(10\,\mathrm{GeV}\right)^2}\left(\frac{10\,\mathrm{keV}}{M}\right), \tag{6.17}$$

when it annihilates with the non-relativistic DM in the halo. For neutrino energies $E_\nu \simeq E_\nu^*$ the scattering cross section is resonantly enhanced, and thus the neutrino flux on Earth should exhibit a distinctive dip at the neutrino energy $E_\nu^*$, if DM couples to neutrinos. In contrast, for a $t$-channel resonance there is no distinctive dip in the neutrino spectrum since such scatterings do not lead to a resonantly enhanced cross section. However, if the scattering cross-section is sufficiently large, the neutrino signal will still be attenuated in the direction of the galactic center, where the DM density is highest.

Neutrinos reaching Earth with energies greater than $E_\nu \gtrsim 200$ TeV are considered to be of ex­tragalactic origin, while those at lower energies are typically generated in the Earth's atmosphere by incident cosmic rays, and therefore carry no information about DM–neutrino interactions. The IceCube collaboration measured the flux of extra-galactic neutrinos up to PeV energies, without observing any distinctive dip or angular dependence in the incoming neutrino flux. This allows one to place limits on the DM/neutrino couplings, $g \lesssim 0.01 - 1$, for $100\,\mathrm{keV} \lesssim M \lesssim \mathrm{GeV}$ and $1\mathrm{keV} \lesssim m_X \lesssim 100\,\mathrm{GeV}$, depending on the details of the interaction model [439].

In 2018 IceCube detected a 290 TeV neutrino event from an identified source, a distant flaring blazar. For this event to have been observed the neutrino had to cross a large column density of DM between the source and the site of detection, due to both the cosmological and the galactic DM distributions, as well as from the dense DM spike presumably surrounding the blazar. This can then be used to place bounds on DM/neutrino interactions [439].

## 6.4   Positrons and electrons

In this section, we focus on the computation of the flux of electrons[14] and positrons from DM annihilations or decays. Electrons and positrons detected on Earth originate from the local galactic environment and undergo a complex propagation process within that environment. The higher their energy, the closer the sources need to be: as a rule of thumb, 50% (90%) of 1 GeV (1 TeV) positrons come from within 1 kpc, see, e.g., Delahaye et al. (2009) in [440]. Dedicated numerical codes such as `Galprop`, `Dragon`, `Picard`, and `Usine` [441] have been developed and refined to model the propagation of 'ordinary' cosmic rays (CRs), i.e., those of astrophysical origin, and can be adapted for use with DM-produced CRs. DM oriented codes such as `MicrOMEGAs` [126], `MadDM` [128] or `DarkMatters` [442] are often interfaced with the codes above and also efficiently do the job. We highlight the main physics involved in the propagation process and

---

[14]We recall that, as discussed in the introduction of this chapter, *anti*-matter particles are of particular interest for DM searches. However, the flux of electrons from DM annihilation/decay has also been considered as a useful quantity, since the flux of electron from other sources is not overwhelmingly larger than that of positrons (see, e.g., fig. 6.12). Furthermore, a combined electron+positron flux is relevant for the experiments that cannot discriminate between charges. It is therefore pertinent to include electrons in the discussion.

review an approximate semi-analytic solution frequently used in DM indirect detection. This approach allows for efficient and reliable computation of predicted CR fluxes from DM. Comparisons with data can be performed either using the approximate solutions or with the dedicated numerical codes mentioned earlier, though it is crucial to recognize that significant uncertainties may exist, as addressed later in this section. While our discussion is mainly centered on electrons and positrons, we will occasionally introduce a more general formalism for future reference.

The DM-induced differential $e^\pm$ flux[15] per unit of energy at any spatial point $\boldsymbol{x}$ and time $t$ is given by $d\Phi_{e^\pm}(t, \boldsymbol{x}, E)/dE = v_{e^\pm} f/4\pi$, with units $(\mathrm{GeV} \cdot \mathrm{cm}^2 \cdot \mathrm{s} \cdot \mathrm{sr})^{-1}$. Here, $v_{e^\pm}$ is the velocity of electrons/positrons, essentially equal to the speed of light $c$ in the regime of weak-scale DM. The $e^\pm$ number density per unit energy, $f(t, \boldsymbol{x}, E) = dn_{e^\pm}/dE$, obeys the diffusion-loss equation in galactic cylindrical coordinates

$$\frac{\partial f}{\partial t} - \nabla\left(\mathcal{K}(E, \boldsymbol{x})\nabla f\right) + \frac{\partial}{\partial z}\left(\mathrm{sign}(z)\, f\, V_{\mathrm{conv}}\right) - \frac{\partial}{\partial E}\left(b_{e^\pm}(E, \boldsymbol{x})f + v_{e^\pm}^2 \mathcal{K}_p(E, \boldsymbol{x})\frac{\partial f}{\partial E}\right) = Q(E, \boldsymbol{x}). \quad (6.18)$$

This equation governs the evolution of the $e^\pm$ number density with an injection rate $Q$, taking into account the various propagation processes that will be discussed shortly. It is derived from the full general formalism that provides a detailed description of CR transport in the galaxy (for classic references, see [443]). It is somewhat simplified by retaining only the terms most relevant to $e^\pm$. Furthermore, since steady state conditions hold, the first term in eq. (6.18) vanishes, and the dependence on time disappears. The other terms are as follows.

▷ DM DM annihilations or DM decays provide the spatially dependent **source term** $Q$ in eq. (6.18),

$$Q = \begin{cases} \dfrac{1}{2}\left(\dfrac{\rho}{M}\right)^2 \langle\sigma v\rangle_{\mathrm{tot}} \dfrac{dN_{e^\pm}}{dE} & \text{(DM annihilation)}, \\[12pt] \dfrac{\rho}{M}\Gamma_{\mathrm{tot}} \dfrac{dN_{e^\pm}}{dE} & \text{(DM decay)}, \end{cases} \quad (6.19)$$

where the $dN_{e^\pm}/dE$ spectra are given in eq. (6.1), and $\rho(\boldsymbol{x})$ is the DM density. That is, DM acts as a diffuse source of $e^\pm$, encompassing the entire DM halo. The intensity is higher at the center and diminishes towards the periphery, modulated by $\rho$ (in the case of DM decays) or $\rho^2$ (in the case of DM annihilations).

▷ **The diffusion coefficient** function $\mathcal{K}$ describes the 'random walk' diffusion of $e^\pm$ as these move in the inhomogeneous turbulent magnetic fields in the galaxy. The diffusion coefficient depends on the momentum $p$ of the propagating electron or positron, and thus also on its energy $E$, being very crudely proportional to the gyro-radius $\sim p/eB$, where $B$ is the average magnetic field.[16] While in principle $\mathcal{K}$ may vary with position $\boldsymbol{x}$, since the distribution of diffusive inhomogeneities changes throughout the galactic halo, the mapping of such variations remains substantially unknown. For example, the diffusion properties could differ inside and outside the galactic arms, as well as inside and outside the galactic disk, subject to the poorly understood local galactic geography. Moreover, including spatial dependence in $\mathcal{K}$ would make the semi-analytic methods much more difficult to implement numerically. Therefore, it is customary to treat $\mathcal{K}$ as a scalar and to disregard the spatial dependence, resulting in $\nabla\left(\mathcal{K}(E)\nabla f\right) = \mathcal{K}(E)\nabla^2 f$.

---

[15]With the notation $e^\pm$ we always refer to the independent fluxes of electrons $e^-$ or positrons $e^+$, which share the same formalism, and not to their sum (for which we use the notation $e^+ + e^-$ when needed). The $e^\pm$ fluxes and the $e^+ + e^-$ flux induced by DM differ by a factor 2, up to possible C violations.

[16]More refined descriptions of the galactic magnetic field are possible, for instance including a regular component in addition to the turbulent one. See e.g. Kachelriess and Semikoz (2019) in [443]. This leads to a more complex description of diffusion, in general not isotropic.

Several models have been proposed to describe the dependence of the diffusion coefficient $\mathcal{K}$ on energy $E$. The simplest parameterisation, routinely used in earlier studies, is given by

$$\mathcal{K}(E) = \mathcal{K}_0 \, v_{e^\pm} \, (E/\,\mathrm{GeV})^\delta, \tag{6.20}$$

where $\mathcal{K}_0$ is a normalization factor and $\delta \sim 0.5 - 1$ is a diffusion spectral index. More recently, a refined parameterization reflecting the observed spectral breaks in CR fluxes [444] has been adopted. When generalized to any charged CR with charge $Ze$, and formulated in terms of rigidity $R \equiv p/Ze$ rather than energy, it takes the form

$$\mathcal{K}(R) = v^\eta \mathcal{K}_0 \left(\frac{R}{1\,\mathrm{GV}}\right)^\delta \left[1 + \left(\frac{R_l}{R}\right)^{(\delta-\delta_l)/s_l}\right]^{s_l} \left[1 + \left(\frac{R}{R_h}\right)^{(\delta-\delta_h)/s_h}\right]^{-s_h}. \tag{6.21}$$

Here, the parameters $R_{l(h)}$ indicate the positions of the spectral breaks, $\delta_{l(h)}$ denote the diffusion spectral indices in the low- and high-rigidity regime, respectively, and $s_{l(h)}$ characterises the rate at which the spectral change takes place around $R_{l(h)}$. In the relativistic $e^\pm$ limit, and ignoring the spectral breaks, the formalism simplifies to the original parameterisation in eq. (6.20). The values of all these parameters will be the focus of a dedicated discussion below.

▷ The $\mathcal{K}_p$ term in eq. (6.18) accounts for the so-called **reacceleration**, i.e., the process in which the propagating particles hit the diffusion centers (i.e., the knots of a turbulent galactic magnetic field) and undergo a second-order Fermi acceleration. That is, the diffusion centers, which are drifting at a certain speed, cause the particles to gain energy. Spatial diffusion thus also leads to a modification of the energy of the propagating cosmic ray and therefore to a *diffusion in momentum space*. One way to parametrise the diffusive re-acceleration coefficient is

$$\mathcal{K}_p(E, \boldsymbol{x}) = \frac{4}{3} \frac{V_a^2}{\mathcal{K}(E, \boldsymbol{x})} \frac{p^2}{\delta(4 - \delta^2)(4 - \delta)}, \tag{6.22}$$

where $V_a$ is an effective Alfvénic speed, a property of the magnetic turbulence centers, and $\delta$ is the spectral index of spatial diffusion, introduced earlier. The dependence of $\mathcal{K}_p$ on the inverse of the spatial diffusion coefficient $\mathcal{K}$ encodes the fact that the more efficiently cosmic rays diffuse in space, the fewer collisions occur, thereby weakening the energy diffusion process.

▷ The $V_{\mathrm{conv}}$ term represents a **convective wind**, assumed to be constant and directed outward from the galactic plane. This wind exerts a force on charged cosmic rays, pushing them away from the galactic plane. Its impact is higher on low energy particles ($E \lesssim$ few GeV) due to their lower rigidity. More refined scenarios with $V_{\mathrm{conv}}$ that is not constant, but rather increases or decreases with $z$, have also been considered.

▷ The $e^\pm$ **energy loss coefficient function** $b_{e^\pm}(E, \boldsymbol{x})$ quantifies the energy losses experienced by the $e^\pm$ during their propagation. In general, it depends on the position $\boldsymbol{x}$, reflecting the fact that energy losses are influenced by the specific environment. It also depends on the energy $E$ of the propagating $e^\pm$. Various mechanisms contribute to these losses, including energy dissipation through Coulomb interactions and ionization with the interstellar gas, bremsstrahlung on the same gas, Inverse Compton Scattering (ICS) on the InterStellar Radiation Field (ISRF), and synchrotron emission in the galactic magnetic field. Schematically, the total energy loss can be expressed as

$$b_{e^\pm}(E, \boldsymbol{x}) \equiv -\frac{\mathrm{d}E}{\mathrm{d}t} = b_{\mathrm{Coul+ioniz}} + b_{\mathrm{brem}} + b_{\mathrm{ICS}} + b_{\mathrm{syn}}. \tag{6.23}$$

The processes above are listed in order of increasing relevance as the $e^\pm$ energy increases. For

detailed expressions of these terms and further insights into their underlying physics, the reader is referred to the relevant literature [425, 445][17], while here we just sketch their main features:

- *Coulomb interactions and ionizations of the gas* are primarily relevant for $e^\pm$ with energies $E \lesssim 1$ GeV.[18] They occur on neutral and ionized galactic gas, with Thompson cross section $\sigma_T = 8\pi r_e^2/3$, where $r_e = \alpha/m_e$ is the classical electron radius, inversely proportional to the electron mass $m_e$. The energy losses are proportional to the density of target atoms and molecules, necessitating detailed spatial maps that are often deduced from external astrophysical observations. The energy loss rate $b_{\text{Coul+ioniz}}$ due to these processes is roughly independent of the $e^\pm$ energy.

- *Bremsstrahlung interactions* are relevant in the range $1$ GeV $\lesssim E \lesssim 10$ GeV. Like Coulomb interactions and ionizations, they occur on neutral and ionized galactic gas too. The relevant cross section contains again the Thompson cross section and an additional factor of $\alpha$ associated to the bremsstrahlung photon emission. The energy loss rate $b_{\text{brem}}$ is linearly dependent on $E$.

- *ICS processes* are dominant for $10$ GeV $\lesssim E \lesssim 100$ GeV. They occur on the three main components of the ISRF: optical star-light, dust-diffused infrared light and the CMB, the latter being a black body spectrum with temperature $T = 2.725$ K. For the optical and infrared components, detailed maps are required for a precise description. The energy dependence of ICS losses is described by $b_{\text{ICS}} \propto E^2$, valid in the non-relativistic Thomson regime.[19] For sufficiently high electron energy, the IC scattering enters the relativistic regime, where the $\gamma e^\pm$ Klein-Nishina cross section becomes smaller than in the Thomson limit. Consequently, a multiplicative factor $R_i^{\text{KN}}(E) < 1$ must be introduced (see, e.g., fig. 2 of Meade et al. (2010) in [445], and eq. (6.24) below), leading to a reduction in $b_{\text{ICS}}(E)$.

- *Synchrotron emission* becomes the dominant channel of energy loss for high-energy electrons and positrons. This process depends on the intensity of the galactic magnetic field at $\boldsymbol{x}$, a quantity that is rather uncertain [446]. Generally speaking, in spiral galaxies like the Milky Way, the strength of the total magnetic field is expected to be of the order of $10$ $\mu$G $-$ $1$ mG and to follow the spiral pattern. Measurements specific to the Milky Way indicate a value around $5-6$ $\mu$G at the position of the Sun and $10-40$ $\mu$G close to the Galactic Center. The intensity of the galactic magnetic field is typically modeled with a profile that peaks at the galactic center and decreases with smooth exponents in both the radial and vertical directions. More detailed profiles, which take into account the galactic morphology of arms and voids, are also sometimes used. The energy dependence of synchrotron emission losses is given by $b_{\text{syn}} \propto E^2$.

The summary result for the function $b_{e^\pm}(E, \boldsymbol{x})$ is illustrated in fig. 6.7. In the figure, the $E^2$ behavior at intermediate energies is apparent. At low energies, the dependence softens, since the Coulomb, ionization and bremsstrahlung losses dominate. At high energies, the dependence also

---

[17]The reader can also consult section 6.7, where some of these same processes are considered for the emission signals they produce.

[18]These ranges are only indicative: the specific details depend on the local density of gas and radiation, as well as the intensity of the magnetic field where the interactions happen, and their relative weight in the impact for energy loss. These ranges are roughly correct for the environment around the solar system.

[19]The Thomson regime in electron-photon Compton scattering is characterized by the condition $\epsilon'_{\text{max}} = 2\gamma\epsilon < m_e$, where $\epsilon$ denotes the energy of the impinging photon, $\epsilon'$ the same quantity in the rest frame of the electron, while $\gamma$ is the Lorentz factor of the electron. When $e^\pm$ scatter on CMB photons ($\epsilon \approx 2\ 10^{-4}$ eV) this condition is satisfied up to $\sim$ TeV $e^\pm$ energies. For scatterings on more energetic starlight ($\epsilon \approx 0.3$ eV), the condition breaks down already above $e^\pm$ energies of $\approx$ few GeV.

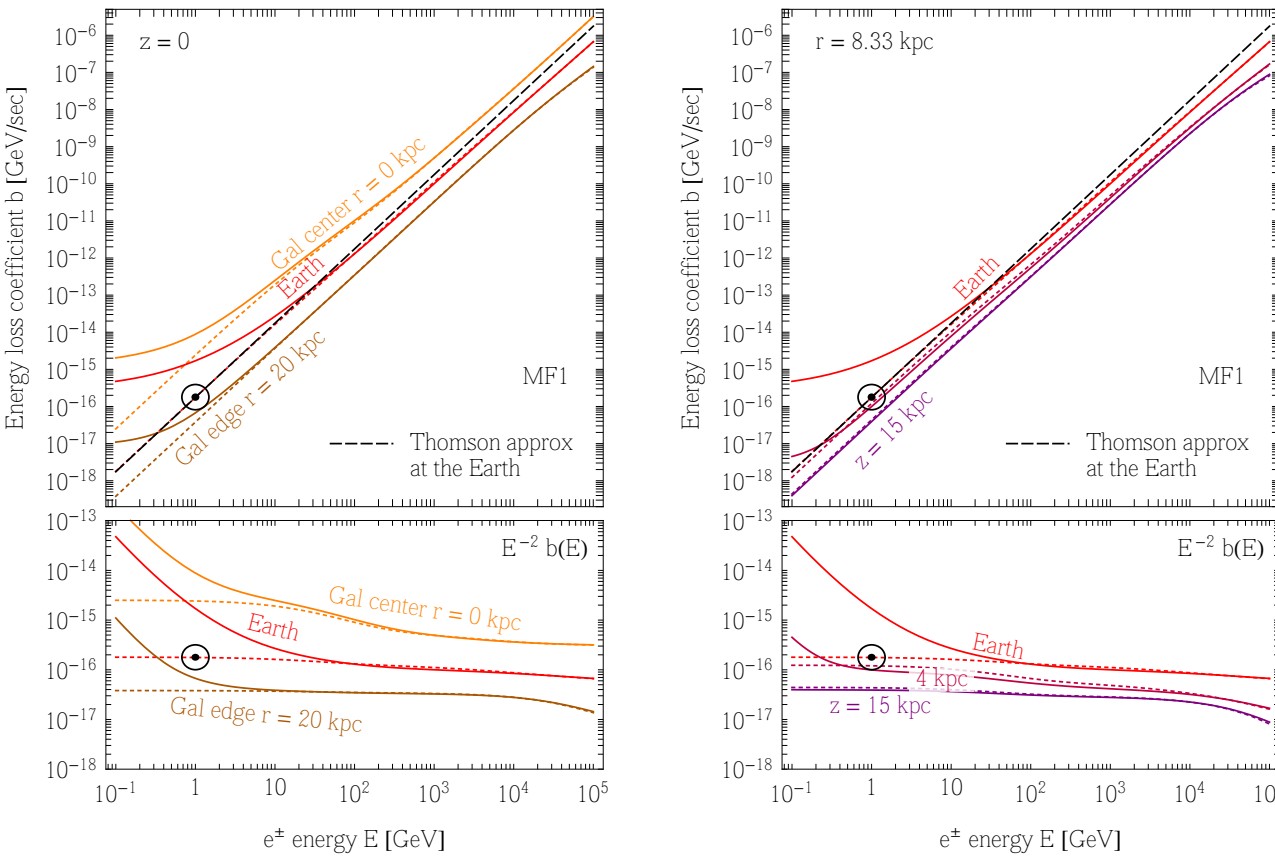

Figure 6.7:  ***Energy loss coefficient function*** *for electrons and positrons in the Milky Way. Left panel: in the galactic disk ($z = 0$), at several locations along the galactic radial coordinate $r$. Right panel: above (or below) the location of the Earth along the coordinate $z$. Here MF1 denotes a choice for the galactic magnetic field. The circled dot identifies the constant value $\tau_\odot$ sometimes adopted. The dotted colored lines are the same function neglecting the low energy losses (essentially Coulomb, ionization and bremsstrahlung), and the black dashed line corresponds to adopting the Thomson approximation (i.e. neglecting the Klein-Nishina factor) for the ICS losses. The lower panels show the same function divided by $E^2$, to highlight the deviations from a simple $E^2$ behaviour. Figure adapted from Buch et al. (2015) in [445].*

softens because of the Klein-Nishina effect. The transition happens earlier at the GC, where starlight is more abundant, and later at the periphery of the Galaxy, where the CMB is the dominant background. At the GC, the dependence eventually re-settles onto an $E^2$ slope at very high energies, where synchrotron losses dominate.

The low-energy losses are often neglected when the focus is on energetic $e^\pm$ originating from weak-scale DM. In such cases, the energy loss can be approximated by

$$b_{e^\pm}(E, \boldsymbol{x}) \approx b_{\text{ICS}}(E, \boldsymbol{x}) + b_{\text{syn}}(E, \boldsymbol{x}) = \frac{4\sigma_{\text{T}}}{3m_e^2} E^2 \tilde{u}, \qquad \tilde{u} = \sum_i u_\gamma(\boldsymbol{x}) R_i^{\text{KN}}(E) + u_B(\boldsymbol{x}), \qquad (6.24)$$

where $u_B = B^2/2$ is the energy density in galactic magnetic fields of intensity $B$ and $u_{\gamma,i} = \int dE_\gamma\, n_i(E_\gamma)$ is the energy density in light, summed over the three main components of the ISRF with photon number

densities $n_i$. An even further approximation can be made by neglecting the spatial dependence in $b_{e\pm}$ and ignoring the Klein-Nishina suppression in the ICS processes. This leads to

$$b_{e\pm}(E) \approx \frac{E^2}{\text{GeV } \tau_\odot} \equiv b_{\text{T}}(E), \tag{6.25}$$

where $\tau_\odot = 5.7 \ 10^{15}$ sec is characteristic of the $e^\pm$ energy losses at the location of the solar system, in this rather drastically simplified scenario.

In semi-analytic solution methods, which are also employed by most of the numerical codes referenced earlier, eq. (6.18) is solved within a specific diffusive region. This region is assumed to be a flat cylinder with the galactic plane at its symmetry plane, and has a height of $2L$ in the $z$ direction and a radius $R = 20 \, \text{kpc}$ in the $r$ direction [440].[20] The location of the solar system is given by the coordinates $(r_\odot, z_\odot) = (8.33 \, \text{kpc}, 0)$. The cylindrical region represents the space where charged particles are presumed to remain confined by the magnetic field, and its shape is inspired by (mostly radiowave) observations of external galaxies. Mathematically, this means that the boundary conditions are imposed in such a way that the $e^\pm$ density $f$ vanishes on the surface of the cylinder, outside of which electrons and positrons propagate freely and escape.

Stepping back, our extensive discussion above highlights that the propagation of electrons and positrons is a complex process, influenced by numerous astrophysical phenomena. Specifically, the propagation relies on several *propagation parameters* present in the preceding equations. These parameters, in order of their introduction, are: $\eta, \mathcal{K}_0, \delta, R_l, \delta_l, s_l, R_h, \delta_h, s_h, V_a, V_{\text{conv}}$, and $L$. Some of these parameters have a large impact on the resulting fluxes of electrons and positrons (as well as anti-protons and anti-nuclei, discussed in the following sections). For instance, the height of the diffusive cylinder $L$ determines how tall is the confinement region. Intuitively, a larger $L$ leads to more extensive confinement, resulting in a higher final flux, especially as positrons originating from as far as the dense regions close to the GC remain confined, and thus can potentially be detected on Earth. Conversely, some parameters exert minimal influence or are strictly constrained by other observational data.[21]

A significant branch of CR research over past decades has been dedicated to determining these parameters using a combination of 'ordinary' CR data and models (i.e., not related to DM), particularly through secondary-to-primary CR ratios (conventionally, the most used is the boron-to-carbon (B/C) ratio). More precisely, one first identifies a *propagation scheme*, i.e., a version of the full transport equation (6.18), either including or omitting individual components, and subsequently identifies plausible ranges for the retained propagation parameters. For concreteness, we will follow the state-of-the-art analysis by Génolini et al. (2019) in [440], which takes advantage of the high-precision CR data provided by the Ams-02 experiment. They selected three propagation schemes termed `BIG`, `SLIM` and `QUAINT`:

▶ `BIG` is the most complete and versatile: it includes a double-break diffusion coefficient as in eq. (6.21), as well as convection and re-acceleration, with all the corresponding parameters.

▶ `SLIM` is a minimalistic version of the former: it does not include convection nor re-acceleration (i.e. $V_{\text{conv}} = V_a = 0$) and fixes $\eta = 1$.

▶ `QUAINT` includes convection and re-acceleration but does not include a low-energy spectral break.

---

[20]This configuration is referred to as the *two-zone diffusion model*: one zone is the diffusive cylinder, the other the thin galactic disk containing the gas (see below).

[21] Additionally, there are interdependences between the parameters describing the astrophysical ingredients. For instance, the thickness $L$ of the CR diffusive halo is related to the vertical extent of the galactic magnetic field discussed on page 205. However, such correlations are often neglected for simplicity, at least in the DM-related CR literature.

|       | $\delta$ | $\mathcal{K}_0$ in kpc$^2$/Myr | $R_l$ in GV | $\delta_l$ | $L$ in kpc |
|-------|----------|---------------------------------|-------------|------------|------------|
| MIN   | 0.509    | 0.01945                         | 4.21        | $-1.450$   | 2.56       |
| MED   | 0.499    | 0.03631                         | 4.48        | $-1.110$   | 4.67       |
| MAX   | 0.490    | 0.06607                         | 4.74        | $-0.776$   | 8.40       |

Table 6.1: ***Propagation parameters*** *for charged particles in the Galaxy, for the* `SLIM` *propagation scheme. The values of the fixed parameters, for this scheme, read* $\eta = 1, s_l = 0.05, R_h = 237$ *GV,* $\delta_h = \delta - 0.19, s_h = 0.04, V_{\mathrm{conv}} = V_a = 0$. *The values are from Génolini et al. (2021) in [440], where equivalent sets for other propagation schemes are given.*

For each of these schemes, specific parameter sets were identified, with table 6.1 showing parameters for the `SLIM` configuration as an example. These sets either maximize or minimize the resulting fluxes of DM-related cosmic rays at Earth.[22] They are thus named MIN, MED, and MAX. Other similar sets have been considered in the past. Works by Donato et al. (2004) and Delahaye et al. (2008) in [440] introduced the original versions of the MIN-MED-MAX sets, now updated by those in table 6.1. Additionally, alternative sets have been proposed by, among others, Di Bernardo et al. (2010), Di Bernardo et al. (2013) and Trotta et al. (2011) in [440].

In summary, setting the parameters in eq. (6.18) to their values in table 6.1 (or to other estimates), and then solving this equation either through semi-analytic or numerical methods, one can calculate the DM annihilation/decay induced fluxes of electrons and positrons at Earth, after their propagation through the Galaxy.

## 6.4.1   Semi-analytic solution via halo functions

It is instructive to examine an explicit solution method within a simplified context. Let us choose the simplest version of the diffusion coefficient, eq. (6.20), neglect the terms for convection and re-acceleration and adopt the simplified form of the energy loss function $b_{\mathrm{T}}(E)$ as in eq. (6.25). Under these assumptions, it is possible to demonstrate [440] that the solution to eq. (6.18) can be expressed as

$$\frac{d\Phi_{e^\pm}}{dE}(E, \boldsymbol{x}) = \frac{1}{4\pi} \frac{v_{e^\pm}}{b_{\mathrm{T}}(E)} \begin{cases} \dfrac{1}{2}\left(\dfrac{\rho_\odot}{M}\right)^2 \displaystyle\int_E^M dE_{\mathrm{s}} \, \langle\sigma v\rangle_{\mathrm{tot}} \dfrac{dN_{e^\pm}}{dE}(E_{\mathrm{s}}) \, \mathcal{I}(\lambda_D(E, E_{\mathrm{s}}), \boldsymbol{x}) & \text{(DM annihilation)}, \\[4mm] \left(\dfrac{\rho_\odot}{M}\right)\displaystyle\int_E^{M/2} dE_{\mathrm{s}} \, \Gamma_{\mathrm{tot}} \dfrac{dN_{e^\pm}}{dE}(E_{\mathrm{s}}) \, \mathcal{I}(\lambda_D(E, E_{\mathrm{s}}), \boldsymbol{x}) & \text{(DM decay)}. \end{cases}$$
(6.26)

That is, the solution can be written as a simple convolution between the spectra at production $dN_{e^\pm}/dE$ and a dimension-less *halo function* $\mathcal{I}(\lambda_D, \boldsymbol{x})$ that depends on a single quantity

$$\lambda_D = \lambda_D(E, E_{\mathrm{s}}) = \sqrt{4\mathcal{K}_0\tau_\odot[E^{\delta-1} - E_{\mathrm{s}}^{\delta-1}]/(1-\delta)},$$
(6.27)

which represents the diffusion length of $e^\pm$ injected with energy $E_{\mathrm{s}}$ and detected with energy $E$. The halo functions essentially act as Green's functions from a source with fixed energy $E_{\mathrm{s}}$ to any energy $E$. They obey the differential equation $\nabla^2\mathcal{I} - (2/\lambda_D)\partial\mathcal{I}/\partial\lambda_D = 0$, derived from eq. (6.18), to be solved with $\mathcal{I} = 0$ on the boundary of the cylinder and the initial condition $\mathcal{I}(0, \boldsymbol{x}) = [\rho(\boldsymbol{x})/\rho_\odot]^p$ (with $p = 2$ for DM annihilations, and $p = 1$ for DM decays). Examples of solutions are shown in fig. 6.8: the halo functions

---

[22]Note that these parameters do not necessarily correspond to the same extremal fluxes in other locations of the Galaxy. For instance, the MAX set leads to the most suppressed flux of positrons in some areas near the GC, while MIN and MED yield the most enhanced one. This variability stems from the differing relative significance of the propagation parameters, which varies between the Earth's vicinity and other galactic locations.

can exceed 1 if containment within the galactic magnetic field enables the collection of $e^{\pm}$ from denser and more distant regions toward the Galactic Center (GC), thereby enhancing the fluxes.

Reverting to the complete form of the energy loss function, eq. (6.24), one can generalize the above formalism and compute *generalized halo functions* $I(E, E_s, \boldsymbol{x})$, which serve the same purpose as the $\mathcal{I}$ functions (see [425] and Buch et al. (2015) in [445]). For the full technical details, we refer the reader to the relevant literature. It is worth emphasizing that the 'halo function formalism' has limitations, as it fails to capture the full complexity of galactic propagation, particularly at low energies where specialized codes are required. Nevertheless, it remains a practically convenient method, enabling the calculation of propagated cosmic ray fluxes through a straightforward integration over the source energy. The halo functions $\mathcal{I}$ encapsulate all the astrophysical considerations (with a distinct halo function $\mathcal{I}$ for each choice of DM distribution profile and $e^{\pm}$ propagation parameters) and are independent of the specific particle physics model. Convoluted with the injection spectra, they give the final spectra after galactic propagation.

The final step in obtaining spectra that are directly comparable to observational data consists of taking into account the effect of **solar modulation** [447]. The solar wind and the solar magnetic field affect significantly the trajectory of charged cosmic rays as they approach the solar system, particularly those with energies below a few GeV. The process is treated effectively, in the so-called *force field approximation*, where it is assumed that solar activity reduces the energy and momentum of charged cosmic rays. The energy spectrum $d\Phi_{\oplus}/dK_{\oplus}$ of charged particles reaching the Earth with kinetic energy $K_{\oplus}$ and momentum $p_{\oplus}$ (known as the Top of the Atmosphere or 'ToA' flux) is approximately related to their energy spectrum in the interstellar medium, $d\Phi/dK$, as

$$\frac{d\Phi_{\oplus}}{dK_{\oplus}} = \frac{p_{\oplus}^2}{p^2}\frac{d\Phi}{dK}, \qquad K = K_{\oplus} + |Ze|\phi_{\text{Fisk}}, \qquad p^2 = 2mK + K^2, \tag{6.28}$$

where $m$ and $Z$ denote the mass and charge of the CR particles. The Fisk potential $\phi_{\text{Fisk}}$ parameterizes the kinetic energy loss in this effective formalism. Its value depends on the activity of the Sun, which is known to follow an 11-year cycle, and can be measured in several ways, including using neutron monitors. Typical values for $\phi_{\text{Fisk}}$ lie between 0.5 GV and 0.9 GV but may range over $0.3 - 1.3$ GV. For instance, $\phi_{\text{Fisk}} \approx 0.5$ GV is characteristic of a minimum of the solar cyclic activity (as observed in the second half of the '90s, late 2000s, and around 2020), whereas $\phi_{\text{Fisk}} \gtrsim 0.9$ GV is typical of a maximum (the early 2000's and the first half of the 2010's), see Ghelfi et al. (2017) in [447] for a historical overview spanning nearly 70 years. More refined treatments, and dedicated codes, have been developed [447], in particular to take into account possible charge dependencies of the process.

## 6.4.2 Positrons and electrons: practical results

The above formalism enables the computation of electron and positron fluxes following their galactic propagation, starting from the injection fluxes presented in section 6.1 and depicted, e.g., in fig. 6.3 (left panel). The **broad features** of the resulting fluxes are:

- The end-point of the spectra remains fixed at $E = M$ (for DM annihilations) or $E = M/2$ (for DM decays). However, in cases of strong re-acceleration, a tail of $e^{\pm}$ with $E > M$ ($E > M/2$) may emerge.

- The overall normalization of the fluxes is governed by the annihilation cross section $\langle \sigma v \rangle$ (or the decay rate $\Gamma$) and is proportional to the DM density squared $\rho^2$ for DM annihilations (to $\rho$ for DM decays).

- Galactic propagation modifies the spectral shape of the fluxes, mostly by redistributing high-energy $e^{\pm}$ to lower energies, though in the case of strong re-acceleration the redistribution to

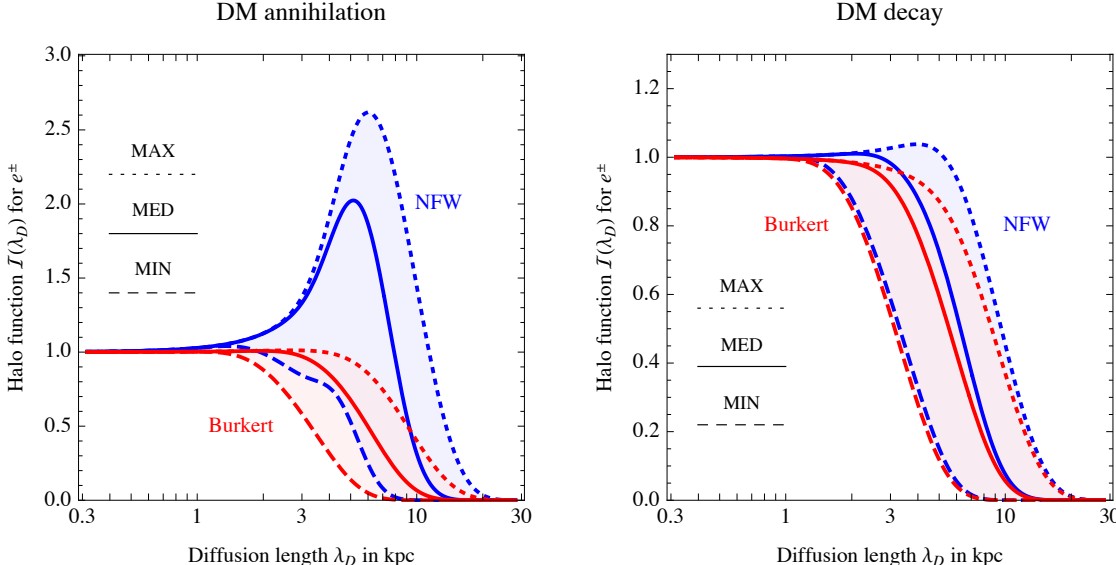

Figure 6.8: **Halo functions** *for $e^{\pm}$ at Sun's location for the propagation parameters in table 6.1 and two DM profiles from table 2.3.*

higher energies is also possible. The distinctive sharp features characteristic of leptonic channels (see fig. 6.3, left panel) are therefore partially washed out. Nevertheless, general characterizations such as "hard spectrum from leptonic channels" versus "soft spectrum from hadronic/gauge boson channels" remain intact.

The processes of injection and propagation introduce also notable sources of **uncertainty** in the predicted fluxes. These uncertainties can be broadly classified into three categories:

▷ *Galactic propagation.* This category is defined by sets of parameters like those listed in table 6.1. They represent the current range of sensible possibilities, encompassing the minimal to maximal fluxes predicted from DM that are compatible with ordinary CR physics. That is, the allowed values of these parameters serve as an estimate of the present uncertainty related to propagation. For $e^{\pm}$, the impact is energy-dependent. Low-energy $e^{\pm}$ (e.g., $E \sim 1 - 10$ GeV) originate from a large volume and are therefore significantly affected by the propagation processes: the span between the MIN and MAX predictions is typically a factor of a few. High-energy $e^{\pm}$, being more localized, have less uncertain flux. In addition, solar modulation, which modifies the CR fluxes at low energies, is subject to large uncertainties.

▷ *DM distribution.* Uncertainties in the distribution of DM within the Galaxy influence the predicted fluxes. For $e^{\pm}$ the impact is energy dependent. For high-energy $e^{\pm}$, only the local value of the DM density is relevant. For low-energy $e^{\pm}$, the whole DM profile is important, including its behavior at the galactic center: the variation in the predicted flux from a cored to a peaked profile is typically a bit less than an order of magnitude for annihilating DM. Furthermore, DM over-densities, if present, can boost the flux in the DM annihilation scenario (see section 6.8.1).

▷ *Energy losses.* The densities of interstellar gas and ISRF as well as the intensity of the magnetic field enter as external ingredients in the propagation formalism described above. The associated uncertainties then translate into the predicted CR fluxes. Their detailed impact is location- and energy-dependent, given that the environmental conditions are more uncertain in the inner regions of the Galaxy than they are locally. Roughly, changing the gas and light densities and the magnetic field within currently accepted ranges can alter the final DM $e^{\pm}$ spectra by a factor of $\sim 2$.

A summary of the data and its implications for DM will be presented in section 6.13.

## 6.5 Anti-protons

The propagation of anti-protons[23] $\bar{p}$ through the galaxy shares many similarities with that of electrons and positrons discussed in the previous section. However, there are also notable qualitative differences, which can be summarized as follows:

i) Energy losses are much less relevant for $\bar{p}$, leading to better preservation of the spectral shape of the injection fluxes (see below for a detailed discussion). On the other hand, different DM annihilation channels tend to give similar $\bar{p}$ spectra, see section 6.1. Moreover, for a given energy, $\bar{p}$ have a longer survival time in the diffusive halo compared to $e^{\pm}$, resulting in a larger collection volume.

ii) Anti-protons are subject to nuclear physics interactions *(spallations)* with the interstellar gas. These interactions modify their flux by removing particles (e.g., through $\bar{p}$ annihilation on interstellar hydrogen) or adding particles (e.g., through spallations that produce additional $\bar{p}$).

The number density of anti-protons per unit energy $f(t, \boldsymbol{x}, K) = dN_{\bar{p}}/dK$, expressed in terms of the kinetic energy $K = E - m_p$, which is used instead of the total energy $E$,[24] obeys a diffusion equation analogous to eq. (6.18) [440, 443]:

$$\frac{\partial f}{\partial t} - \boldsymbol{\nabla} \cdot (\mathcal{K}(K, \boldsymbol{x}) \boldsymbol{\nabla} f) + \frac{\partial}{\partial z} \left( \text{sign}(z) \, f \, V_{\text{conv}} \right) - \frac{\partial}{\partial K} \left( b_{\bar{p}}(K, \boldsymbol{x}) f + v_{\bar{p}}^2 \, \mathcal{K}_p(K, \boldsymbol{x}) \frac{\partial f}{\partial K} \right) =$$
$$= Q(K, \boldsymbol{x}) - 2h \, \delta(z) \left( \Gamma_{\text{ann}} + \Gamma_{\text{non-ann}} \right) f. \tag{6.29}$$

We refer to the previous section for the common terms, and only comment on the differences.

▷ The **source term** $Q$ due to DM DM annihilations or DM decay has a form analogous to eq. (6.19), with $E$ replaced by $K$ and with the $\bar{p}$ spectra from eq. (6.1). DM acts as a diffuse source of $\bar{p}$, encompassing the entire halo.

▷ The pure **diffusion term** is again assumed to be space independent. Its simplest parametrization reads, analogously to eq. (6.20),

$$\mathcal{K}(K) = v_{\bar{p}} \, \mathcal{K}_0 \, (p/\, \text{GeV})^{\delta}, \tag{6.30}$$

where $p = [K^2 + 2m_p K]^{1/2}$ and $v_{\bar{p}} = [1 - m_p^2/(K + m_p)^2]^{1/2}$ are the anti-proton momentum and speed. A more refined parametrization, which includes spectral breaks, is as in eq. (6.21).

▷ The **reacceleration** term is analogous to eq. (6.22). Other terms contributing to reacceleration, that can appear at higher orders, are neglected here.

▷ The **energy losses**, expressed by the function $b_{\bar{p}}(K, \boldsymbol{x})$, are qualitatively different for anti-protons in comparison to $e^{\pm}$. Since $m_p \gg m_e$, the ICS and synchrotron losses, which scale as $1/m^2$, are

---

[23]We recall that, as discussed in the introduction of this chapter, anti-matter particles are of greater interest for DM searches. CR protons from astrophysical sources are so overwhelmingly dominant ($p/\bar{p} \approx 10^5$ at GeV energies, see, e.g., fig. 6.12), that the protons are practically never used for DM searches. Therefore, CR protons are not considered here.

[24]This distinction is not particularly relevant for fluxes originating from TeV-scale DM, i.e., at energies much larger than the proton mass $m_p$, but it is important for the low energy tails and in the case of small DM masses.

negligible. In fact, all of the energy losses are not very important for anti-protons and are generally neglected. However, in detailed studies one should still include [448]

$$b_{\bar{p}}(K, \boldsymbol{x}) = b_{\text{Coul+ioniz}} + b_{\text{adia}}, \tag{6.31}$$

where the two terms represent:

– *Coulomb interactions and ionizations* on the interstellar gas, analogous to those for $e^{\pm}$. These processes are limited to the galactic disk, where the gas is present.[25] Therefore, the $b_{\bar{p}}$ term includes a factor $2h\,\delta(z)$, which effectively localizes the rate to the $z = 0$ plane. Here, $h \approx 100$ pc $\ll L$ is the thickness of the galactic disk. These energy loss processes are still proportional to the Thomson cross section and the number density of gas atoms, but the detailed expressions are somewhat different from those for $e^{\pm}$ (see, e.g., the summary in Strong and Moskalenko (1998) in [448]).

– *Adiabatic losses.* The convective processes encoded in the $V_{\text{conv}}$ term induce a loss of energy: as the CR are 'pushed' by the convective wind, their density decreases, i.e., they can be modeled as an expanding gas. Without external heat exchange, they then lose energy due to decreasing pressure. This leads to $b_{\text{adia}} = V_{\text{conv}}p^2/3hE$.

▷ The last term in eq. (6.29) is a **sink term** that removes $\bar{p}$ from the flux. It accounts for various nuclear physics interactions of anti-protons with the interstellar gas,[26] which is localized in the galactic disk, hence the factor $2h\,\delta(z)$.

– The first contribution describes the annihilations of $\bar{p}$ on interstellar protons in the galactic plane, with the rate $\Gamma_{\text{ann}} = (n_{\text{H}} + 4^{2/3}n_{\text{He}})\sigma_{\bar{p}p}^{\text{ann}}v_{\bar{p}}$, where $n_{\text{H}} \approx 1/\text{cm}^3$ is the hydrogen density, $n_{\text{He}} \approx 0.07\,n_{\text{H}}$ is the Helium density (the factor $4^{2/3}$ accounts in an effective way for the differences in geometrical cross-sections). The cross section $\sigma_{\bar{p}p}^{\text{ann}}$ has been studied in some detail in the literature [449], with the parametrization introduced by Tan and Ng (1983) still widely used.

– The second contribution, similarly, describes the interactions on interstellar protons in the galactic plane in which the $\bar{p}$'s do not annihilate but lose a significant fraction of their energy. In a technical sense, these particles should remain in the flux with degraded energy and can indeed be considered a form of energy loss, see, e.g,. Boudaud et al. (2015) in [448]: they are referred to as "tertiary anti-protons". It is common, however, to simplify the treatment by considering thrm removed from the flux altogether. Consequently, the form of this term is similar to the previous one, with the appropriate cross section $\sigma_{\bar{p}p}^{\text{non-ann}}$. In the literature, one often finds the expression for the sum $\sigma_{\bar{p}p}^{\text{inel}} = \sigma_{\bar{p}p}^{\text{ann}} + \sigma_{\bar{p}p}^{\text{non-ann}}$ [425, 449].

– An additional re-injection term $Q^{\text{tert}}$ for the 'tertiary anti-protons' discussed in the previous bullet point could be included but is typically neglected and we do not show it in eq. (6.29). If included, it would take the form of a convolution of the non-annihilating cross section with the spectrum $f(K)$, thereby transforming eq. (6.29) into an integro-differential equation. Solving this modified equation would be more complex; one would have to first obtain a solution without the tertiary source term, and then iteratively update it using the revised source term from the previous step, see Donato et al. (2001) in [449]. The phenomenological impact of including $Q^{\text{tert}}$ is usually limited, and thus neglecting it is warranted.

---

[25]The (inverse) bremsstrahlung process, where an impinging anti-proton hits an atomic electron at rest, causing the emission of a photon, can also be considered. However, it does not lead to appreciable energy loss for the anti-proton and is therefore neglected.

[26]In contrast, the interactions of $e^{\pm}$ with the gas are negligible and were not included in eq. (6.18), see Gaggero et al. (2013) in [445] for a quantitative assessment.

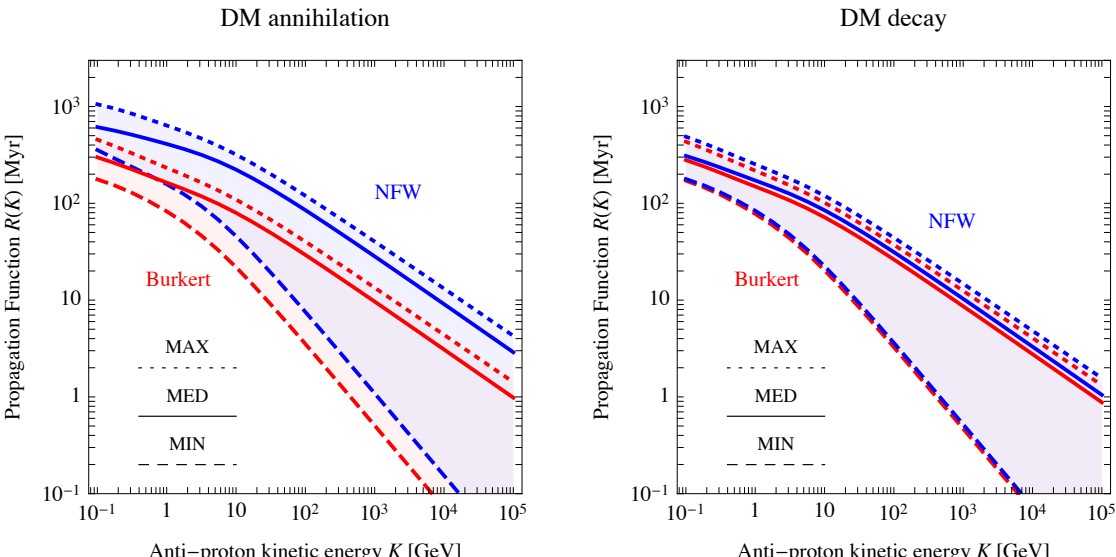

Figure 6.9: ***Propagation functions*** $\mathcal{R}(K)$ *for $\bar{p}$ following from the propagation parameters in table 6.1 and the two DM profiles in table 2.3, for DM annihilations (left) and DM decays (right).*

Similar to the treatment of positrons and electrons, eq. (6.29) is solved under steady-state conditions ($\partial f/\partial t = 0$), within a cylindrical confining region of thickness $2L$. It is assumed that the anti-protons that reach the boundaries escape, and consequently, their density at the edges vanishes. The methodology for determining the propagation parameters entering in the various parts of the equation involves fitting ordinary CR data, akin to the approach used for $e^{\pm}$. The propagation schemes discussed in the previous section and the parameters listed in table 6.1 apply to $\bar{p}$ too. The resulting solution yields the flux of anti-protons from DM annihilations or decays after their galactic propagation. Solar modulation is incorporated as the final phase of the propagation process, as elaborated on page 209.

### 6.5.1 Semi-analytic solution via propagation functions

It is illuminating to examine an explicit solution within a simplified context. Let us adopt the simple version of the diffusion coefficient, eq. (6.30), and neglect all the $\bar{p}$ energy losses, as well as the tertiary re-injection component. In this approximation, eq. (6.29) can be semi-analytically solved, and the anti-proton differential flux at the position of the Earth, $d\Phi_{\bar{p}}/dK = v_{\bar{p}}f/(4\pi)$, is given by [450]

$$\frac{d\Phi_{\bar{p}}}{dK}(K, \boldsymbol{r}_{\odot}) = \frac{v_{\bar{p}}}{4\pi} \begin{cases} \dfrac{1}{2}\left(\dfrac{\rho_{\odot}}{M}\right)^2 \langle\sigma v\rangle_{\text{tot}} \, \mathcal{R}(K) \, \dfrac{dN_{\bar{p}}}{dK} & \text{(DM annihilation)}, \\[3mm] \left(\dfrac{\rho_{\odot}}{M}\right)\Gamma_{\text{tot}} \, \mathcal{R}(K) \, \dfrac{dN_{\bar{p}}}{dK} & \text{(DM decay)}. \end{cases} \tag{6.32}$$

The 'propagation functions' $\mathcal{R}(K)$ are independent of the particle physics model (described by the $dN_{\bar{p}}/dK$ energy spectra) and encode all the astrophysics of $\bar{p}$ production and propagation. Distinct propagation functions are defined for DM annihilations and for DM decays, each still depending on the choice of DM galactic profile, and on the selection of propagation parameters, such as those listed in table 6.1. The functions $\mathcal{R}(K)$ for representative choices of these parameters are shown in fig. 6.9.[27]

---

[27]The difference with previously published propagation functions at low kinetic energy (e.g., in [425]) is due to the absence of convection in the currently adopted parameters.

### 6.5.2   Anti-protons: practical results

The formalism described above allows to compute the anti-proton fluxes after galactic propagation, starting from the injection fluxes detailed in section 6.1 and depicted, e.g., in fig. 6.3 (middle panel). Analogous to the treatment of electrons and positrons, it is insightful to examine qualitatively the resulting broad features of the $\bar{p}$ fluxes.

- ○ The end-point of the spectra remains at $E = M$ for DM annihilations ($E = M/2$ for DM decays). However, in presence of a strong re-acceleration a tail of $\bar{p}$ with $E > M$ may emerge. It should be noted that the end-point of the $\bar{p}$ spectra is typically not a significant feature, since the spectra commonly peak at $K \approx 10^{-2}M$.

- ○ The overall normalization of the fluxes is governed by the annihilation cross section $\langle \sigma v \rangle$ (or the decay rate $\Gamma$) and is also proportional to the DM density squared $\rho^2$ for DM annihilations ($\rho$ for DM decays).

- ○ Owing to the minimal impact of energy losses and the relatively limited influence of spallations (which are confined to the thin disk), the galactic propagation of $\bar{p}$ typically maintains the spectral shape of the fluxes. However, as commented above, the $\bar{p}$ spectra across different channels are rather self-similar, and thus do not carry much information.

As with electrons and positrons, the injection and propagation processes introduce sources of **uncertainty** in the predicted $\bar{p}$ fluxes.

- ▷ Galactic propagation is characterized by sets of parameters such as those in table 6.1. The variation between MIN and MAX predictions is roughly independent on the $\bar{p}$ energy and typically does not exceed half an order of magnitude. This uncertainty was reduced considerably recently (propagation uncertainty bands of up to almost two orders of magnitudes were considered in the past), thanks to improved determinations of the propagation parameters (see, e.g., Génolini et al. (2021) in [440], and references therein). Solar modulation further alters the fluxes at low energies.

- ▷ The uncertainty in the galactic DM distribution affects the predicted fluxes, roughly independently of energy. The difference between the predicted flux for a cored profile and a peaked profile is typically a factor of a few for annihilating DM, and much smaller for decaying DM. In addition, if DM sub-halos are present, they boost the flux in the case of DM annihilations (see section 6.8.1).

- ▷ Uncertainties from the spallation cross section and other interactions are generally considered minor for the DM $\bar{p}$ flux and are not typically evaluated. However, they may be important for an accurate computation of the astrophysical background flux.

The limited impact of these uncertainties on the DM $\bar{p}$ fluxes, combined with the fact that the astrophysical background $\bar{p}$ spectrum is also rather well under control (compared, for example, to positrons) has made anti-protons a probe of choice for investigating DM and producing competitive bounds, see section 6.13.2 for the current status.

## 6.6   Anti-deuteron and heavier anti-nuclei

The propagation of DM generated antideuterons and other light nuclei (essentially anti-helium) through the Galaxy follows closely that of anti-protons, section 6.5, with a few modifications [451]. The transport equation remains as given in eq. (6.29). Diffusion, dictated by the electromagnetic properties of the particles, is identical for anti-nuclei and anti-protons, with the simple substitution that the deuteron mass $m_d$ or helium mass $m_{^3\mathrm{He}}$, $m_{^4\mathrm{He}}$ must replace the proton mass in the expressions for the kinetic

energy and momentum. The convection, diffusive re-acceleration and energy loss terms are the same as those in the previous section.[28] The source term features the spectra discussed in section 6.1.1.

The treatment of spallations of $\bar{d}$ and $\overline{\text{He}}$ on interstellar gas (both 'annihilating' and 'non-annihilating' reactions) is less straightforward than for $\bar{p}$, largely due to the scarcity of experimental nuclear data on $\bar{d}$ and $\overline{\text{He}}$. The values for $\sigma_{\bar{d}p}^{\text{ann}}$, $\sigma_{\bar{d}p}^{\text{non-ann}}$ and $\sigma_{\bar{d}p}^{\text{inel}}$, as well as the equivalent cross sections for the $\overline{\text{He}}\,p$ processes, are required: these can be obtained experimentally from the measurements of the charge conjugated reactions induced by the $d\bar{p}$ scattering, and partially also from the $p\bar{p}$ scattering data. They can also be estimated from more general formulas derived for cosmic ray nuclei. Often, a good approximation is already to adopt simple scaling relations, such as $\sigma_{\bar{d}p}^{\text{inel}} \approx 2\,\sigma_{\bar{p}p}^{\text{inel}}$.

With the ingredients described above, one can compute an anti-nucleon or anti-helium propagation function $R(K/n)$, for both annihilations and decays, in a manner analogous to the procedure followed for anti-protons. This computation can be carried out for any chosen Dark Matter (DM) galactic profile and for any set of propagation parameters listed in table 6.1. It then becomes straightforward to calculate the anti-deuteron and anti-helium differential flux at Earth's location, using a formalism identical to the one in eq. (6.32). Finally, solar modulation can be applied in a manner similar to that discussed for anti-protons, incorporating the appropriate kinematic quantities as detailed in eq. (6.4.1).

## 6.7   Secondary photons

As discussed above, the galactic $e^{\pm}$ generated by DM in the diffusion volume of the Galaxy lose essentially all their energy into photons by means of two processes: inverse Compton scattering and synchrotron radiation. Bremsstrahlung radiation also plays a significant role in regions close to the galactic disk, where gas is abundant, particularly for electrons in the energy range of $\sim 1 - 10\,\text{GeV}$.

The fluxes of ICS (and bremsstrahlung) $\gamma$ rays generated in this way, as well the microwave synchrotron radiation, can thus be signatures of DM in the Milky Way. The same emissions can be searched for also in other systems, such as in dwarf galaxies, clusters of galaxies and even in the cosmological flux, provided that the necessary environmental conditions are met: the presence of ambient light for ICS (with the understanding that the CMB is always present), gas for bremsstrahlung, and a magnetic field for synchrotron emission. Collectively, these emissions are referred to as *secondary photons*, in contrast to the prompt photons discussed in section 6.2.

The ICS flux has been investigated in great detail in recent years. One of its distinguishing features is that it originates from 'everywhere' within the diffusion volume of the galactic halo, including regions where the astrophysical background is reduced (e.g., at high latitudes). Moreover, in areas where synchrotron energy losses are sub-dominant in comparison to ICS energy losses, the resulting ICS $\gamma$ flux is subject to only moderate astrophysical uncertainties, owing to energy conservation. The microwave synchrotron emission, on the other hand, is generated in significant quantities near the GC, where the intensity of the magnetic field and the density of DM are highest, resulting in greater uncertainty and more background. However, it can also originate from large latitudes. Finally, bremsstrahlung emission may be a relevant signature of DM under certain conditions (such as large gas density and $e^{\pm}$ energy $\lesssim 10$ GeV). In any case, it must be taken into account, if one aims at consistently modeling the other emissions with which it competes.

In this section, we provide a detailed description of the computation of ICS emission (section 6.7.1), bremsstrahlung emission (section 6.7.2), and synchrotron radiation (section 6.7.3). The methods that we describe build on the standard electro-magnetic formalism [443, 445] and use tools in the form of

---

[28] For low-$Z$ nuclei, it is customary to use the kinetic energy per nucleon, $K/n$, as a variable. While the total kinetic energy $K$ of the particle must be used in the equations, the spectra are often presented in terms of $K/n$.

*generalized halo functions* for ICS, bremsstrahlung and synchrotron, as initially introduced in [425] and Buch et al. (2015) in [445]. These functions, which are analogous to the halo functions used for $e^\pm$ propagation, are computed once and for all. They then enable the computation of the emission through a simple convolution with the injected electron spectrum, making the phenomenology faster to analyze for any DM model.

## 6.7.1   Inverse Compton gamma rays

The ICS signal has its origin in the following process, alluded to earlier: energetic $e^+$ and $e^-$, produced in the halo of the Galaxy by the annihilations or decays of DM particles, hit the low energy photons of the ambient bath (the CMB, infrared light and starlight). These collisions constitute an inverse Compton scattering, upscattering the ambient photon energy from its initial low value $E_\gamma^0$ to a final value of up to $E_\gamma \approx 4\gamma^2 E_\gamma^0$. Here $\gamma = E_e/m_e$ is the relativistic factor of the electrons and positrons. As a rule of thumb, a 1 TeV electron will produce a $\sim 1.5$ GeV soft $\gamma$-ray when scattering off the CMB ($E_\gamma^0 \approx 10^{-4}$ eV), or a $\sim 150$ GeV hard $\gamma$-ray when scattering off UV starlight ($E_\gamma^0 \approx 10$ eV).

An accurate description of the ambient radiation field is important to compute the ICS emission in a reliable way. A recent appraisal of the radiation maps can be obtained from [452]. It displays modifications of at most a factor of a few in the intensity of the IR and starlight fields with respect to previous determinations. This variation can be considered as an estimate of the astrophysical uncertainty related to the environment, affecting at the same time the computation of ICS energy losses and the ICS signal itself. It is worth mentioning, however, that these uncertainties may partly offset each other: an increase in ambient light density results in greater energy loss for the propagating $e^\pm$, leading to a suppressed $e^\pm$ spectrum, but at the same time it enhances ICS emission. In regions where ICS is the dominant energy-loss/emission process for the electrons and positrons, the two effects partly compensate each other.

The differential flux of ICS photons from a a solid angle element $d\Omega$ can be written in terms of the emissivity $j(E_\gamma, r)$ of a cell located at a distance $r \equiv |\boldsymbol{x}|$ from the GC as follows:

$$\frac{d\Phi_{\mathrm{IC}\gamma}}{dE_\gamma \, d\Omega} = \frac{1}{4\pi} \frac{1}{E_\gamma} \int_{\mathrm{l.o.s.}} ds \, j(E_\gamma, r(s, \theta)). \tag{6.33}$$

For a radiative process, in general terms, the emissivity is derived as a convolution of the spatial density of the emitting medium with the power that it radiates (see e.g. Rybicki and Lightman (1979) in [443]). In this case

$$j(E_\gamma, r) = 2 \int_{m_e}^{M(/2)} dE_e \, \mathcal{P}_{\mathrm{IC}}(E_\gamma, E_e, r) \, \frac{dn_{e^\pm}}{dE_e}(r, E_e), \tag{6.34}$$

where $\mathcal{P} = \sum_i \mathcal{P}_{\mathrm{IC}}^i$ is the differential power emitted into photons due to ICS radiative processes (the sum runs over the different components of the photon bath). The electron (or positron) number density post-diffusion and energy losses, $dn_{e^\pm}/dE_e$, has been computed in subsection 6.4. (Note that this quantity was represented as $f$ in that section for simplicity, see page 203; $dn_{e^\pm}/dE_e$ just corresponds to eq. (6.26), removing the $v_{e^\pm}/4\pi$ factor.) The minimal and maximal electron energies in the integration limits in eq. (6.34) are fixed by the electron mass $m_e$ and the mass of the DM particle $M$. The '/2' notation applies to the decay case. The overall factor of 2 before the integral accounts for the fact that an equal population of positrons, in addition to electrons, is produced by DM annihilations/decays, and both contribute to the emission.

The radiated power $\mathcal{P}_{\mathrm{IC}}$, using the full Klein-Nishina expression for $e\gamma$ scattering, is (we refer the

reader to [443, 445] and references therein for more details on the derivation)

$$\mathcal{P}_{\mathrm{IC}}^i(E_\gamma, E_e, \boldsymbol{x}) = \frac{3\sigma_{\mathrm{T}}}{4\gamma^2} \int_{1/4\gamma^2}^1 dq \left( E_\gamma - \frac{E_\gamma}{4q\gamma^2(1-\epsilon)} \right) \times$$
$$\times \frac{n_i\big(E_\gamma^0(q), \boldsymbol{x}\big)}{q} \left[ 2q \ln q + q + 1 - 2q^2 + \frac{1}{2}\frac{\epsilon^2}{1-\epsilon}(1-q) \right]. \tag{6.35}$$

The integration variable is

$$q = \frac{\epsilon}{\Gamma_E(1-\epsilon)}, \qquad \text{where} \qquad \Gamma_E = \frac{4E_\gamma^0 E_e}{m_e^2} \qquad \text{and} \qquad \epsilon = \frac{E_\gamma}{E_e}, \tag{6.36}$$

and lies in the range $1/4\gamma^2 \approx 0 \leq q \leq 1$, as expressed by the limits of the integral in eq. (6.35). Correspondingly, the energy $E_\gamma$ lies in the range[29] $E_\gamma^0/E_e \leq E_\gamma \leq E_e\,\Gamma_E/(1+\Gamma_E)$. The non-relativistic (Thompson) limit corresponds to $\Gamma_E \ll 1$, so that $\epsilon \ll 1$, and therefore the last addend in the integrand of $\mathcal{P}$ can be neglected. At the same time, the integration variable $q$ reduces to $y \equiv E_\gamma/(4\gamma^2 E_\gamma^0)$ with $0 \leq y \leq 1$. Hence, in the Thomson limit,

$$\mathcal{P}_{\mathrm{IC}}^i(E_\gamma, E_e, \boldsymbol{x}) \approx \frac{3\sigma_{\mathrm{T}}}{4\gamma^2} E_\gamma \int_0^1 dy \frac{n_i\big(E_\gamma^0(y), \boldsymbol{x}\big)}{y} \left[ 2y \ln y + y + 1 - 2y^2 \right] \qquad \text{[Thomson limit]}. \tag{6.37}$$

By substituting $\mathcal{P}_{\mathrm{IC}}$ and $n_{e\pm}$ in eq. (6.34), the IC differential flux can be recast in a more convenient form:

$$\frac{d\Phi_{\mathrm{IC}\gamma}}{dE_\gamma\, d\Omega} = \frac{1}{E_\gamma^2} \frac{r_\odot}{4\pi} \begin{cases} \dfrac{1}{2}\left(\dfrac{\rho_\odot}{M}\right)^2 \displaystyle\int_{m_e}^M dE_{\mathrm{s}} \, \langle\sigma v\rangle_{\mathrm{tot}} \dfrac{dN_{e\pm}}{dE}(E_{\mathrm{s}})\, I_{\mathrm{IC}}(E_\gamma, E_{\mathrm{s}}, b, \ell) & \text{(annihilation)}, \\[3mm] \dfrac{\rho_\odot}{M} \displaystyle\int_{m_e}^{M/2} dE_{\mathrm{s}} \, \Gamma_{\mathrm{tot}} \dfrac{dN_{e\pm}}{dE}(E_{\mathrm{s}})\, I_{\mathrm{IC}}(E_\gamma, E_{\mathrm{s}}, b, \ell) & \text{(decay)}, \end{cases} \tag{6.38}$$

where $E_{\mathrm{s}}$ is the $e^\pm$ injection energy and $I_{\mathrm{IC}}(E_\gamma, E_{\mathrm{s}}, b, \ell)$ (which has the dimensionality of an energy) is a halo function for the IC radiative process. Thus this formalism permits to express the flux of ICS $\gamma$ in terms of the convolution of the electron injection spectrum $dN_{e\pm}/dE$ and the appropriate halo functions, in close analogy with the formalism for charged particles. Explicitly, we can write $I_{\mathrm{IC}}$ in terms of the ICS ingredients discussed above and the generalized halo functions for $e^\pm$ introduced in section 6.4.1:

$$I_{\mathrm{IC}}(E_\gamma, E_{\mathrm{s}}, b, \ell) = 2\, E_\gamma \int_{\mathrm{l.o.s.}} \frac{ds}{r_\odot} \left( \frac{\rho(r(s,\theta))}{\rho_\odot} \right)^p \int_{m_e}^{E_{\mathrm{s}}} dE\, \frac{\sum_i \mathcal{P}_{\mathrm{IC}}^i(E_\gamma, E, r(s,\theta))}{b_{e\pm}(E, r(s,\theta))}\, I(E, E_{\mathrm{s}}, r(s,\theta)), \tag{6.39}$$

where $p = 1(2)$ for DM decay (annihilation). The intensity of the interstellar radiation, $\sum_i n_i$, cancels out in the ratio $\sum_i \mathcal{P}_i/b_{e\pm}$, up to the sub-leading synchrotron contribution. The final step in obtaining the differential ICS $\gamma$ flux $d\Phi_{\mathrm{IC}\gamma}/dE_\gamma d\Omega$ consists of performing the convolution integral over $E_{\mathrm{s}}$ in eq. (6.38) for a particular DM generated prompt $e^\pm$ energy spectrum.

The differential flux from a region $\Delta\Omega$ is obtained by integrating over the galactic polar coordinates $b$ and $\ell$, defined in eq. (6.6),

$$\frac{d\Phi_{\mathrm{IC}\gamma}}{dE_\gamma} = \iint db\, d\ell \, \cos b \, \frac{d\Phi_{\mathrm{IC}\gamma}}{d\Omega\, dE_\gamma}. \tag{6.40}$$

Due to the intertwined dependence of the halo functions on $b$ and $\ell$ and on $E_\gamma$ and $E_s$, the geometrical

---

[29]The combination $E_e\Gamma_E$ reproduces the 'rule of thumb estimate' for $E_\gamma$ mentioned at the beginning of the section.

integral cannot be factored out as in the case of prompt $\gamma$ rays. Consequently, a simple definition of the $\bar{J}$ factor is not possible.

We should emphasize that the halo function formalism is specific to the galactic signal. For other systems, or for more complex analyses, the ICS signal can be computed directly using the ingredients in eq.s (6.33)$-$(6.36).

## 6.7.2   Bremsstrahlung gamma rays

The bremsstrahlung signal is generated by energetic electrons and positrons, produced in the galactic halo by the annihilations or decays of DM particles. The $e^\pm$ interact with the electromagnetic field of the atoms constituting interstellar gas, emitting a bremsstrahlung photon. The energy of the emitted photon peaks at a fraction of the initial energy of the $e^\pm$, typically between $1/10$ and $1/2$, depending on both the $e^\pm$ spectrum and on local conditions. Notably, this process becomes the dominant channel for energy loss of $e^\pm$ in the energy range of $1 - 10$ GeV, particularly in regions where the density of interstellar gas is sizeable.

For an accurate assessment of the bremsstrahlung signal, one requires maps of the gas distribution and composition within the Galaxy, which are only partially available [453]. On large scales, coarse-grained maps for the density of hydrogen (in atomic, molecular and ionized form) as well as atomic helium are available: they show an average density of $\mathcal{O}(1)$ particles/cm$^3$ in the galactic disk, rapidly decreasing in the vertical direction, e.g., to $\mathcal{O}(0.01)$ atoms/cm$^3$ at 1 kpc above the galactic plane. The gas density in the Galaxy's central regions is expected to be substantially higher, possibly reaching $\mathcal{O}(100)$ particles/cm$^3$ within the inner 200 pc around the Galactic Center (the so-called Central Molecular Zone), or even higher in the inner few parsecs, where accretion occurs on the central black hole. These variations in density introduce significant astrophysical uncertainties in the computation of the DM bremsstrahlung signal. However, analogous to the ICS case, these uncertainties are partially mitigated by the conservation of energy. A higher gas density results in increased energy loss for propagating $e^\pm$, leading to a suppressed spectrum but simultaneously enhancing bremsstrahlung emission. In regions where bremsstrahlung dominates as the energy-loss/emission process for $e^\pm$, these two effects largely compensate each other.

The computation of the bremsstrahlung emission and its generalized halo functions follows quite closely the computation for ICS in the previous subsection. For completeness, we briefly summarize the essential components here. In exact analogy with eq. (6.38), the bremsstrahlung differential flux can be written as:

$$\frac{d\Phi_{\mathrm{brem}\gamma}}{dE_\gamma\, d\Omega} = \frac{1}{E_\gamma^2}\frac{r_\odot}{4\pi} \begin{cases} \dfrac{1}{2}\left(\dfrac{\rho_\odot}{M}\right)^2 \displaystyle\int_{m_e}^{M} dE_{\mathrm{s}}\, \langle\sigma v\rangle_{\mathrm{tot}} \dfrac{dN_{e^\pm}}{dE}(E_{\mathrm{s}})\, I_{\mathrm{brem}}(E_\gamma, E_{\mathrm{s}}, b, \ell) & \text{(annihilation)}, \\[4mm] \dfrac{\rho_\odot}{M}\displaystyle\int_{m_e}^{M/2} dE_{\mathrm{s}}\, \Gamma_{\mathrm{tot}} \dfrac{dN_{e^\pm}}{dE}(E_{\mathrm{s}})\, I_{\mathrm{brem}}(E_\gamma, E_{\mathrm{s}}, b, \ell) & \text{(decay)}, \end{cases} \tag{6.41}$$

where now (in analogy with eq. (6.39)) the generalized halo function for bremsstrahlung is

$$I_{\mathrm{brem}}(E_\gamma, E_{\mathrm{s}}, b, \ell) = 2\, E_\gamma \int_{\mathrm{l.o.s.}} \frac{ds}{r_\odot}\left(\frac{\rho(r(s,\theta))}{\rho_\odot}\right)^p \int_{m_e}^{E_{\mathrm{s}}} dE\, \frac{\mathcal{P}_{\mathrm{brem}}(E_\gamma, E, r(s,\theta))}{b_{e^\pm}(E, r(s,\theta))}\, I(E, E_{\mathrm{s}}, r(s,\theta)), \tag{6.42}$$

where as before $p = 1(2)$ for DM decays (annihilations), and $I$ is the $e^\pm$ generalized halo function. The bremsstrahlung power is

$$\mathcal{P}_{\mathrm{brem}}(E_\gamma, E, \boldsymbol{x}) = c\, E_\gamma \sum_i n_i(\boldsymbol{x}) \frac{d\sigma_i(E_\gamma, E)}{dE_\gamma}, \tag{6.43}$$

where $n_i$ are the number densities of the gas species and the differential bremsstrahlung cross section is given by

$$\frac{d\sigma_i(E_{e\pm}, E_\gamma)}{dE_\gamma} = \frac{3}{8\pi}\frac{\alpha\sigma_T}{E_\gamma}\left\{\left[1 + \left(1 - \frac{E_\gamma}{E_{e\pm}}\right)^2\right]\phi_1^i - \frac{2}{3}\left(1 - \frac{E_\gamma}{E_{e\pm}}\right)\phi_2^i\right\}. \tag{6.44}$$

Here, $\phi_{1,2}^i$ are scattering functions, which depend on the properties of the scattering system: ionized, atomic or molecular, hydrogen or helium. We direct the reader to the relevant literature for their explicit forms (see, e.g., Buch et al. (2015) in [445] for a recapitulation and references).

We stress again that the halo function formalism is specific to the galactic signal. For other systems, or for more complex analyses, the bremsstrahlung signal can be computed directly using the power in eq. (6.43), in conjunction with the equivalent of the general expressions in eq.s (6.33)−(6.34).

### 6.7.3 Synchrotron radio waves

Radio waves are produced by the DM generated high-energy electrons and positrons, via the synchrotron process, as they gyrate in the galactic magnetic field. As a rule of thumb, an electron or positron with momentum $p$ moving in a turbulent magnetic field of intensity $B$ generates radio waves that peak at frequency $\nu \approx 3eBp^2/4\pi m_e^3 \approx 4.2\,\text{Hz}\,(B/\mu\text{G})(p/m_e)^2$. For example, an electron with energy 1 GeV (10 GeV) in a 5 $\mu$G magnetic field, which is typical of the local galactic environment, would emit radio waves at 80 MHz (8 GHz). Radio maps of the Milky Way, with varying degrees of coverage and completeness, are available between 22 MHz and 2.3 GHz. In addition, PLANCK-LFI microwave maps cover up to 70 GHz [3].

The synchrotron signal depends on the intensity of the magnetic field, whose uncertainties we have already briefly discussed on page 205. For signals that originate very close to the GC, the impact can be sizable, since in that region the magnetic field is highly uncertain (see, e.g., Bertone et al. (2009) in [454]). In contrast, for signals from the galactic halo, the resulting uncertainty in the radio emission was found to be minimal (see, e.g., Buch et al. (2015) in [445]).

The synchrotron power (in erg s$^{-1}$ Hz$^{-1}$), emitted by an isotropic distribution of relativistic electrons with energy $E_e$ in a uniform magnetic field $B$, is given by

$$\mathcal{P}_{\text{syn}}(\nu, E_e, \alpha) = \sqrt{3}\,\frac{e^3\,B\sin\alpha}{m_e\,c^2}F(x), \tag{6.45}$$

where

$$x = \frac{\nu}{\nu_c'}, \qquad \nu_c' = \frac{\nu_c}{2}\sin\alpha, \qquad \nu_c = \frac{3}{2\pi}\frac{e}{m_ec}B\gamma^2. \tag{6.46}$$

Here, $\alpha$ is the angle between the line of sight and the magnetic field direction, and $\gamma = E_e/m_e$ is the Lorentz factor of the electron or positron. The synchrotron kernel $F(x)$ is defined as

$$F(x) = x\int_x^\infty K_{5/3}(x')dx',$$

where $K_n$ is the modified Bessel function of the second kind of order $n$. When the magnetic field is randomly oriented, as is the case in the galaxy, the synchrotron power must be averaged over the pitch angle $\alpha$, leading to

$$\mathcal{P}_{\text{syn}}(\nu, E_e) = \frac{1}{2}\int_0^\pi d\alpha\,\sin(\alpha)\,\mathcal{P}_{\text{syn}}(\nu, E, \alpha). \tag{6.47}$$

For relativistic electrons ($\gamma \gtrsim 2$) one obtains (see, e.g., Ghisellini et al. (1988) in [445])

$$\mathcal{P}_{\mathrm{syn}}(\nu, E) = 2\sqrt{3}\, \frac{e^3 B}{m_e c^2} y^2 \left[ K_{4/3}(y) K_{1/3}(y) - \frac{3}{5} y \left( K_{4/3}(y)^2 - K_{1/3}(y)^2 \right) \right], \tag{6.48}$$

with $y = \nu/\nu_c$. Integrating this expression over $\nu$ yields the total power emitted by an electron of energy $E$ across all frequencies, i.e., eq. (6.24) (synchrotron portion). The synchrotron emissivity $j_{\mathrm{syn}}(\nu, r)$ is then computed using the equivalent of the general expression in eq. (6.34).

Finally, the observable of interest for comparison with observational data is the intensity $\mathscr{I}$ of the synchrotron emission (in erg cm$^{-2}$ s$^{-1}$ Hz$^{-1}$ sr$^{-1}$) from a certain direction of observation. It is obtained by integrating the emissivity along the line-of-sight. Schematically,

$$\mathscr{I}(\nu, b, \ell) = \frac{1}{4\pi} \int_{\mathrm{l.o.s.}} ds\, j_{\mathrm{syn}}(\nu, r, z), \tag{6.49}$$

where the point in $(r, z)$ is identified by the parameter $s$ along the line of observation, characterized by the galactic latitude $b$ and longitude $\ell$: $r(s, b, \ell), z(s, b, \ell)$.

Collecting the above expressions (equations (6.49) and the equivalent of (6.34), with (6.48) and the generalized version of (6.26)), the synchrotron intensity $\mathscr{I}$, at a fixed frequency $\nu$ and at galactic coordinates $(b, \ell)$, is given by

$$\mathscr{I}(\nu, b, \ell) = \frac{r_\odot}{4\pi} \begin{cases} \dfrac{1}{2} \left( \dfrac{\rho_\odot}{M} \right)^2 \displaystyle\int_{m_e}^{M} dE_s\, \langle \sigma v \rangle_{\mathrm{tot}} \dfrac{dN_{e^\pm}}{dE}(E_s)\, I_{\mathrm{syn}}(\nu, E_s, b, \ell) & \text{(DMannihilation)}, \\[4mm] \left( \dfrac{\rho_\odot}{M} \right) \displaystyle\int_{m_e}^{M/2} dE_s\, \Gamma_{\mathrm{tot}} \dfrac{dN_{e^\pm}}{dE}(E_s)\, I_{\mathrm{syn}}(\nu, E_s, b, \ell) & \text{(DMdecay)}, \end{cases} \tag{6.50}$$

with the *generalized synchrotron halo function* $I_{\mathrm{syn}}(\nu, E_s, b, \ell)$ defined as

$$I_{\mathrm{syn}}(\nu, E_s, b, \ell) = \int_{\mathrm{l.o.s.}} \frac{ds}{r_\odot} \left( \frac{\rho(r, z)}{\rho_\odot} \right)^p 2 \int_{m_e}^{E_s} dE \frac{\mathcal{P}_{\mathrm{syn}}(\nu, E)}{b_{e^\pm}(E, r, z)}\, I(E, E_s, r, z), \tag{6.51}$$

in terms of $I$, the generalized halo function for $e^\pm$. Here, $p = 1(2)$ for the DM decays (annihilations), and, again, $r, z$, are to be written in terms of galactic coordinates $s, \ell, b$.

For completeness, we mention that the synchrotron intensity $\mathscr{I}$ is often traded for the *brightness temperature* $T(\nu)$, expressed in Kelvin, which is just the temperature that a black body would have to possess in order to emit the same intensity at a given frequency. One has [30]

$$T(\nu) = \frac{\mathscr{I}(\nu)}{2\,\nu^2}. \tag{6.52}$$

Once again, we emphasize that the halo function formalism is specific to the galactic signal. For other systems, or when performing more complex analyses, the synchrotron signal can be computed by directly utilizing the power expression in eq. (6.48), along with the generic definitions provided in eq.s (6.34) and (6.49).

## 6.8   Enhancements

The indirect detection signals can be enhanced by a number of different mechanisms. In this section, we discuss the boost caused by clumps in the DM distribution (section 6.8.1) and the enhancements that

---

[30]Here we take $k_B = c = 1$, where $k_B$ the Boltzmann constant.

depend on the micro-physics of the annihilation process (section 6.8.2). These enhancements typically arise only for annihilating DM, whereas in scenarios involving decaying DM, such enhancements are absent.

## 6.8.1   Boost due to DM sub-halos

In the preceding subsections, we assumed that the cosmic and galactic DM distribution was smooth. However, one of the main results of the clustering history is the emergence of a hierarchical structure comprising halos and sub-halos of various sizes throughout the cosmos. This structure gives rise, a fortiori, to the presence of small DM sub-halos within the larger halos of galaxies. The expected number distribution of these sub-structures as a function of their mass was discussed in section 2.2.5.

Small-scale inhomogeneities typically enhance the indirect detection signals of DM annihilations, since these signals are proportional to $\langle \rho^2 \rangle > \langle \rho \rangle^2$ [455]. The position of the DM sub-halos within the galaxy and the cosmos is generally unknown, so that the enhancement is usually taken into account by an overall multiplicative number known as the *boost factor* $B \sim \langle \rho^2 \rangle / \langle \rho \rangle^2$, by which the signal from a smooth DM distribution should be multiplied. In a simplified scenario where DM is uniformly distributed within a volume $V$ and becomes concentrated within a sub-volume $V' < V$, this means that $B = V/V'$.

In principle, the computation of the boost factor $B$ follows in a straightforward manner from the mass function formalism discussed in section 2.2.5. Using the predicted number $n$ of subhalos per mass $\mathcal{M}$, $dn/d\mathcal{M}$, one can include the annihilation signal from each subhalo by integrating over this halo mass distribution, yielding the total signal contributed by substructures. Explicitly, we have

$$B \propto \int_{\mathcal{M}_{\min}} d\mathcal{M} \frac{dn}{d\mathcal{M}} \int dr \ 4\pi r^2 \rho_{\mathrm{sub}}^2(r, \mathcal{M}), \tag{6.53}$$

where $\rho_{\mathrm{sub}}(r, \mathcal{M})$ is the density profile of a sub-halo of mass $\mathcal{M}$ and the coordinate $r$ refers to the center of the subhalo. For DM density $\rho_{\mathrm{sub}}$ we can use one of the forms discussed in section 2.2.1 for the Galaxy, if one assumes, as it is often done, that the subhalos are 'small-size reproductions' of the main halo. However, numerical simulations often reveal that subhalos possess higher DM densities in their inner regions and are, on average, more concentrated than 'normal' halos of the same mass. Therefore, more peaked $\rho_{\mathrm{sub}}$ profiles are sometimes used. In eq. (6.53), the integral over the volume of the subhalo is truncated at a specific radius (typically its virial radius). The integral over the halo masses, in turn, is cut from below at $\mathcal{M}_{\min}$, the minimal mass of the sub-halos (which depends on the microscopic properties of the DM particle as discussed in section 2.2.5). It is noteworthy that the integral is primarily dominated by small halo masses.

However, detailed studies have revealed that the situation is more complex, especially regarding galactic DM signals [456]. First, the halo mass function $dn/d\mathcal{M}$ gets modified during the late stages of galactic evolution. Specifically, sub-halos can be tidally stripped by the gravitational influence of the host halo and other sub-halos, leading to their dissolution. This phenomenon particularly affects small halos, modifying the value of the minimal halo mass. Additionally, sub-halos can be loosened and subsequently dissolved when crossing the galactic disk due to gravitational interactions, an effect referred to as *'disk shocking.'* Encounters with individual stars have also been shown to potentially impact these structures. Cosmological simulations (including baryons) have been used to model the above effects and to compute more realistic boost factors, although the challenge of fully matching the simulated galaxies to the real Milky Way remains. Analytical approaches have been used as well, but they often struggle to capture the full complexity of these processes and often have to adopt simplifying assumptions. In general terms, these gravitational effects are more pronounced in the central regions of the Galaxy, which is thus more severely depleted of subhalos, while the outskirts remain populated. Consequently, the boost factor for $\gamma$-rays from the GC areas is expected to be minimal. That is, in general the boost factor depends on the

observed/sampled area, and is not simply a common overall factor.

Finally, the effective boost factor of the signals detected at Earth depends both on the nature and the energy of the cosmic ray species, i.e., whether one considers $\gamma$-rays (or neutrinos), electrons/positrons or anti-protons/anti-nuclei. This dependence arises because the signal collected at Earth is sensitive to the source volume, as mentioned in the respective sections 6.2.1, 6.4 and 6.5, and this volume, in turn, depends on the particle energy. For instance, the boost factor is higher for high-energy positrons than for low-energy ones: low-energy $e^+$ originate from a large volume encompassing regions towards the GC, where the smooth DM density is high, and therefore the weight of the subhalos is comparatively lower. Conversely, the boost factor is higher for low-energy anti-protons than for high-energy ones: low-energy $\bar{p}$ are more affected by convection and the (subdominant) energy losses and therefore come from a smaller, more local volume, where the component of the signal due to subhalos is comparatively higher than that from the smooth component. As a result, one should consider a variety of energy-dependent boost factors $B_i(E)$, for each species $i$. However, these variations are globally mild and thus the approximation of using a single unified boost factor $B$ remains widely used in practice.

The predictions for the boost factor(s) have varied over the years. Initial works estimated very high values, often in the hundreds or thousands. Values as high as $B \approx 10^9$ have even been suggested. More detailed recent analyses found that boost factors do not exceed $B \sim \mathcal{O}(10)$. The boosts for $e^\pm$ and $\bar{p}$ are found to reach at most $B \approx 30$ (at high and low energy respectively, and in extremal configurations). The boost factors for $\gamma$-rays, while dependent on the abundance of clumps in the observed region as previously discussed, have been found to be similar in magnitude. It is worth noting that non-standard cosmologies, featuring enhanced small-scale primordial fluctuations, phase transitions, or topological defects, may give rise to more DM clumps and consequently larger boost factors. These concepts are further explored in section 4.6 in the context of BH formation.

## 6.8.2   Sommerfeld and bound-state enhancement

In models where an attractive force among two DM particles is mediated by a boson (for example the $W, Z$ vectors, or some dark photon) with mass $M_V$ and coupling $\alpha_g = g^2/4\pi$ to DM, the cross section of DM annihilations is enhanced in the relevant non-relativistic limit, $v_{\rm rel} \ll 1$ [132]. This Sommerfeld enhancement was discussed in section 4.1.5 and is estimated as

$$S \sim \max\left(1, \frac{v_{\max}}{\max(v_{\rm rel}, v_{\min})}\right), \tag{6.54}$$

where $v_{\max} = \pi\alpha_g$ and $v_{\min} = M_V/M$. The Sommerfeld effect enhances both the cross section for cosmological thermal freeze-out (with relative DM velocity $v_{\rm rel}^{\rm cosmo} \sim 1/\sqrt{25}$, see below eq. (4.10)) as well as the cross section for indirect detection ($v_{\rm rel}^{\rm MW} \sim 10^{-3}$ in the Milky Way). Consequently, in DM models where the Sommerfeld effect is present, and assuming thermal freeze-out, the DM annihilation cross-section relevant for indirect detection in Milky Way can be enhanced by a factor of up to $\sim v_{\rm rel}^{\rm cosmo}/v_{\rm rel}^{\rm MW} \sim 200$ compared to its 'standard' value given in eq. (4.13). Indirect signals from DM annihilations in dwarf galaxies may receive even greater enhancements, as $v_{\rm rel}$ is smaller, especially in the smallest dwarfs (see [457]).

The Sommerfeld effect is accompanied by bound-state effects, which can lead to larger resonant enhancements at specific DM velocities. Such enhancements occur when the collision energy equals the mass of a particular bound state. This same effect can arise in models where DM is accompanied by extra particles that can be resonantly produced in DM DM annihilations.

# 6.9    Neutrinos from the Sun

The DM particles in the halo of the Milky Way have a small, yet finite, probability of scattering with a nucleus of a massive celestial body, such as the Sun or the Earth, if their orbit intersects with it. If the DM particles' velocity after scattering is less than the escape velocity from that body, they become gravitationally bound and begin to orbit around it. Upon additional scatterings, they eventually sink into the center of the body and accumulate, building up a local DM over-density concentrated in a relatively small volume. In this dense region, the DM particles can annihilate into Standard Model particles, leading to the production of energetic neutrinos. The neutrinos, after undergoing oscillations and interactions within the dense matter of the astrophysical body, are the only species that can escape. A detection of high-energy neutrinos from the center of the Sun or the Earth (on top of the lower-energy neutrino backgrounds due to nuclear fusion and radioactive processes) would thus constitute a signal of DM [424].

The expected differential neutrino flux is:

$$\frac{d\Phi_\nu}{dE_\nu} = \frac{\Gamma_{\rm ann}}{4\pi d^2}\frac{dN_\nu}{dE_\nu}, \tag{6.55}$$

where $d$ is the distance of the neutrino source from the detector (either the Sun–Earth distance $r_{\rm SE}$ or the Earth radius $R_\oplus$), $dN_\nu/dE_\nu$ is the neutrino flux produced per DM annihilation after taking into account neutrino propagation effects (oscillations and absorption), and $\Gamma_{\rm ann}$ is the DM annihilation rate. The latest results from IceCube and Antares (see section 5.6.8) set bounds on the latter, at levels ranging from $10^{20}$ to $10^{23}$ annihilations/sec, depending on the DM mass and annihilation channel. These bounds can be translated into constraints on the capture rate of DM particles inside the celestial body and therefore on the DM scattering cross sections on nuclei, as discussed below. Neutrinos emanating from the centers of the massive celestial bodies serve as an illustrative example of a DM search strategy that employs the tools of indirect detection to probe the physical quantities involved in standard direct detection.

More involved scenarios can also be considered [458]. For instance, Garani and Palomares-Ruiz (2017) and Maity et al. (2023) considered the case in which DM scatters on electrons rather than nuclei. Zentner (2009) investigated the case in which DM also possesses 'self-interactions', leading to enhanced capture rate. Icecube derived bounds on this scenario (Albuquerque et al. (2014)). Additional scenarios are discussed in section 6.10.1.

In the rest of this section, we restrict our attention to the conventionally considered case of WIMP-like DM particles accumulating and annihilating into Standard Model neutrinos in the center of a celestial body. Our focus will primarily be on the Sun, as the signals from the Earth are not promising for this standard scenario.

## Solar atmospheric neutrinos as a background

An irreducible background to the high energy neutrinos from DM annihilations in the center of the Sun comes from neutrinos produced by cosmic ray interactions within the solar corona [459]. This phenomenon is analogous to the creation of the familiar 'atmospheric neutrinos', but occurs in the 'atmosphere' of the Sun rather than the Earth. When computing this effect, neutrino oscillations must be taken into account. The spectrum of these solar atmospheric neutrinos is harder than the corresponding terrestrial one due to the lower gas densities in the corona, which cause less energy loss for the showering particles, and due to the influence of the intense solar magnetic field. This background is effectively irreducible, as the angular resolution of current neutrino experiments is insufficient to distinguish between the neutrinos produced across the entire surface of the Sun and those originating from the center where DM annihilations occur. According to the latest recalculations, this background is only a factor of 10 below the current sensitivity of the experiments.

## 6.9.1   Computation of the DM annihilation and capture rates

To begin with, let's consider DM after capture. Repeated scatterings thermalise DM particles to the temperature at the center of the Sun, $T_\circledast = 15.5\ 10^6\,\mathrm{K}$, such that the DM density acquires a spherically symmetric Boltzmann form, $n(r) \propto \exp[-M\,\phi(r)/T_\circledast]$, where $\phi(r) = \int_0^r dr\,GM(r)/r^2$ is the Newton potential inside the Sun, written in terms of the solar mass $M(r)$ enclosed within a sphere of radius $r$. Taking for simplicity the matter density in this volume to be constant and equal to the central density of solar matter, $\rho_\circledast = 151\,\mathrm{g/cm^3}$, all integrals can be explicitly evaluated. One finds $\phi(r) - \phi(0) = 2\pi G\rho_\circledast r^2/3$, so that the DM particles are concentrated within a small region around the center of the Sun,

$$n(r) \propto e^{-r^2/r_{\mathrm{DM}}^2} \quad \text{with} \qquad r_{\mathrm{DM}} = \left(\frac{3T_\circledast}{2\pi\,G\rho_\circledast\,M}\right)^{1/2} \approx 0.01\,R_\circledast\sqrt{\frac{100\,\mathrm{GeV}}{M}}, \tag{6.56}$$

where $R_\circledast \approx 7\ 10^8\,\mathrm{m}$ is the radius of the Sun.[31] Similar expressions hold for other astrophysical bodies, after adapting appropriately the $_\circledast$ quantities.[32]

Next, we turn our attention to the calculation of the number $N$ of DM particles captured inside the Sun. This varies with time as

$$\frac{dN}{dt} = \Gamma_{\mathrm{capt}} - \Gamma_{\mathrm{evap}} - 2\Gamma_{\mathrm{ann}}, \tag{6.57}$$

where:

▷ $\Gamma_{\mathrm{capt}}$ is the capture rate discussed below; it does not depend on $N$.

▷ $\Gamma_{\mathrm{evap}}$ is the DM evaporation rate proportional to $N$; it is negligible in the Sun unless DM is lighter than $M \lesssim T_\circledast/v_{\mathrm{esc}} \sim\ \mathrm{GeV}$.

▷ $\Gamma_{\mathrm{ann}}$ is the DM annihilation rate; it is proportional to $N^2$. Since 2 DM particles participate in annihilation, the annihilation rate appears multiplied by 2 in (6.57). It is given by an equation similar to eq. (4.16),

$$2\Gamma_{\mathrm{ann}} = \int dV\,n^2(\boldsymbol{x})\,\langle\sigma v\rangle = C_{\mathrm{ann}}\,N^2, \tag{6.58}$$

where $n(\boldsymbol{x})$ is the number density of DM particles at position $\boldsymbol{x}$ inside the Sun, such that the total number of DM particles is $N = \int dV\,n$. Using the DM density distribution in eq. (6.56) gives $C_{\mathrm{ann}} \sim \langle\sigma v\rangle/r_{\mathrm{DM}}^3$ or, more precisely,

$$C_{\mathrm{ann}} = \langle\sigma v\rangle\left(\frac{GM\rho_\circledast}{3T_\circledast}\right)^{3/2}. \tag{6.59}$$

Neglecting $\Gamma_{\mathrm{evap}}$ and then solving the first order differential equation (6.57) gives

$$\Gamma_{\mathrm{ann}} = \frac{\Gamma_{\mathrm{capt}}}{2}\tanh^2\left(\frac{t}{\tau}\right) \simeq \frac{\Gamma_{\mathrm{capt}}}{2}, \tag{6.60}$$

---

[31]The size of the thermal sphere of DM can also be estimated using the virial theorem, obtaining a similar result. The average kinetic energy of the thermalized DM is $\langle K\rangle = \frac{3}{2}T_\circledast$. The average potential energy of a DM particle of mass $M$ located at the outer shell is $\langle V\rangle = -G\,M(r_{\mathrm{DM}})\,M/r_{\mathrm{DM}}$ with $M(r_{\mathrm{DM}}) = \frac{4}{3}\pi r_{\mathrm{DM}}^3\,\rho_\circledast$, assuming that the DM density is subdominant with respect to the solar matter density. Corrections are necessary when this no longer holds or if DM starts self-gravitating: this does not happen in the Sun, but is conceivable in other astrophysical bodies (see section 6.10.2). The virial theorem $\langle K\rangle = -\frac{1}{2}\langle V\rangle$ gives $r_{\mathrm{DM}} = (9T/4\pi\,G\rho_\circledast M)^{1/2}$.

[32]In this section and throughout this chapter, we use the subscript $_\circledast$ to indicate quantities related to the Sun, differentiating them from those labeled with $_\odot$. For example, $\rho_\circledast$ refers to the Sun's central density, while $\rho_\odot$ is the DM density at the Sun's position in the Galaxy.

where $\tau = 1/\sqrt{\Gamma_{\mathrm{capt}} C_{\mathrm{ann}}}$ is the characteristic time-scale set by the competing processes of capture and annihilation. The last equality in (6.60) holds when the elapsed time is large enough compared to $\tau$ so that one can approximate $\tanh(t/\tau) \approx 1$. This is typically achieved for the Sun, but not for the Earth. Physically, this means that the fast (compared to the age of the Sun) processes of capture and annihilation come to an equilibrium: any additional captured particles thermalize and eventually annihilate away. In this case, the annihilation rate $\Gamma_{\mathrm{ann}}$ is determined entirely by the capture rate $\Gamma_{\mathrm{capt}}$, which acts as the bottleneck.

## The DM capture rate

We can distinguish two regimes for the capture rate of DM by a macroscopic celestial body (the body has radius $R$ and contains $N_N$ matter particles (e.g., hydrogen nuclei) with mass $m_N$):

1. The capture rate is of geometric size, if the DM/matter cross section is large enough, $\sigma_N \gg \sigma_* \approx \pi R^2 / N_N$, so that every DM particle that hits the body also gets captured,

$$\Gamma_{\mathrm{capt}} = \frac{\rho_{\mathrm{DM}}}{M} \langle v_{\mathrm{rel}} \rangle \pi R^2 \approx \frac{1.3 \ 10^{26}}{\mathrm{sec}} \frac{\rho_{\mathrm{DM}}}{0.3 \, \mathrm{GeV/cm^3}} \frac{\mathrm{TeV}}{M} \frac{\langle v_{\mathrm{rel}} \rangle}{10^{-3}} \frac{R^2}{R_\odot^2}, \qquad (6.61)$$

where $\rho_{\mathrm{DM}}/M$ is the number density of DM particles at the location of the body. In the case of the Sun, $\sigma_* \sim 10^{-35} \, \mathrm{cm^2}$. Such a large scattering cross section is experimentally excluded, as mentioned above.

2. If DM particle has only a small probability of interacting with the body, the capture rate is estimated as

$$\Gamma_{\mathrm{capt}} \sim \frac{\rho_{\mathrm{DM}}}{M} \langle v_{\mathrm{rel}} \rangle \ \sigma_N N_N \ \wp_i(v_{\mathrm{rel}}, v_{\mathrm{esc}}), \qquad (6.62)$$

where $\wp$ is a capture probability discussed below. It is significant, if the DM velocity is smaller than the escape velocity from the body, $v_{\mathrm{esc}} \gtrsim v_{\mathrm{rel}} \sqrt{M/m_N}$.

More precisely, the capture rate is given by[33]

$$\Gamma_{\mathrm{capt}} = \frac{\rho_{\mathrm{DM}}}{M} \sum_i \sigma_i \int_0^{R_\odot} dr \ 4\pi r^2 \ n_i(r) \int_0^\infty dv \ 4\pi v^2 \frac{f(v)}{v} (v^2 + v_{\odot \mathrm{esc}}^2) \wp_i(v, v_{\odot \mathrm{esc}}), \qquad (6.63)$$

where $\sigma_i$ is the low-energy DM cross section for DM scattering on nucleus of species $i$, assumed to be isotropic. The celestial body was assumed to be spherical with radius $R_\odot$ and density $\rho_{\mathrm{mat}}(r)$, while the sum in eq. (6.63) runs over all types of nuclei $i$ with masses $m_i$ and mass fractions $\epsilon_i$, such that $n_i = \rho_{\mathrm{mat}}(r)\epsilon_i/m_i$ is their number density and $N_i$ is their total number. The function $f(v)$ is the angular average of the DM velocity distribution in the body's rest frame, neglecting the gravitational attraction of the body itself (it is equal to $f_\odot(v)$ in the case of the Sun, see section 2.3.1). The gravitational pull of the body is taken into account through $v_{\odot \mathrm{esc}}(r)$, the escape velocity at radius $r$, so that $v^2 + v_{\odot \mathrm{esc}}^2$ is the squared velocity of DM when it reaches radius $r$.[34] The probability that a scattering leads to DM capture is given by

$$\wp_i(v, v_{\odot \mathrm{esc}}) = \frac{1}{E_{\mathrm{max}}} \int_{E_{\mathrm{min}}}^{E_{\mathrm{max}}} d\Delta \ |F_i(\Delta)|^2 \approx \max\left(0, \frac{E_{\mathrm{max}} - E_{\mathrm{min}}}{E_{\mathrm{max}}}\right), \qquad (6.64)$$

---

[33]For a slightly more detailed treatment, we refer the reader to Baratella et al. (2014) in [424], and for an exhaustive review to the references in [424].

[34]We neglected the gravitational effect of other bodies in the solar system, a likely justified approximation as per [460].

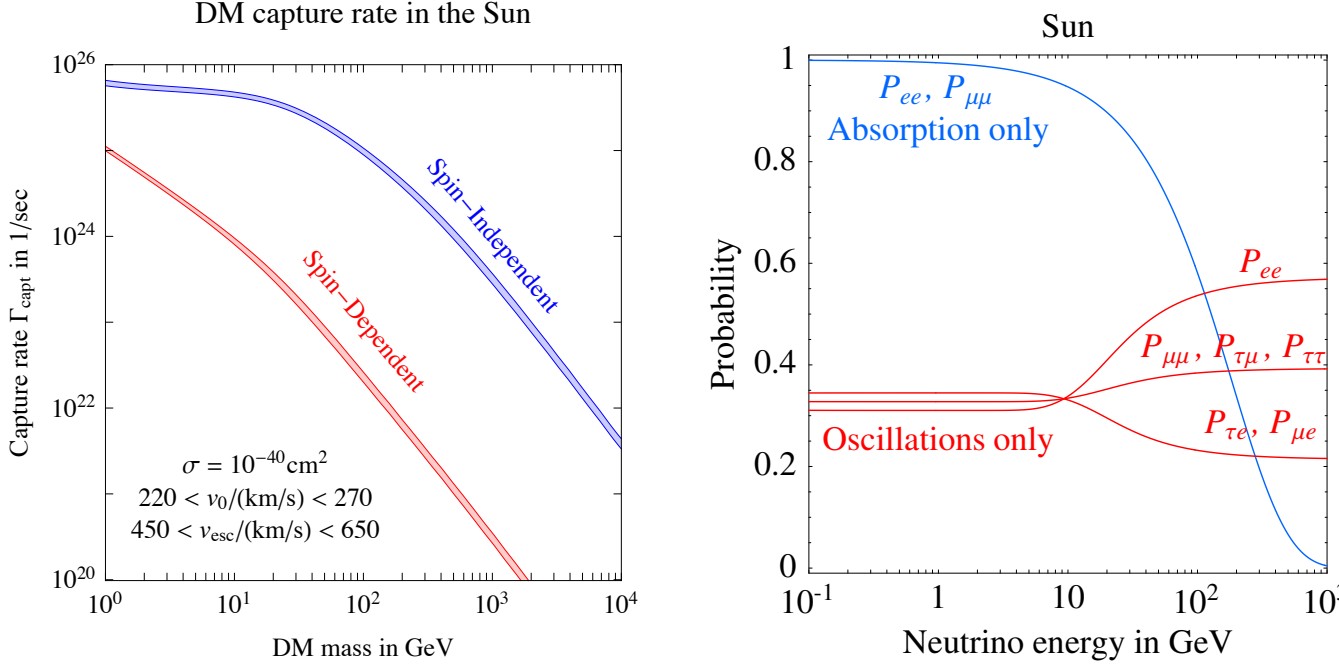

Figure 6.10: Left: ***Capture rate*** *of DM particles in the Sun,* $\Gamma_{\rm capt}$*. The capture rate is linearly proportional to the DM cross-section on a proton, which we take to be* $\sigma = 10^{-40}\,{\rm cm}^2$ *for both SI and SD scatterings.* Right*: Illustration of* ***the effects of oscillations and CC absorption*** *on the neutrino flux produced in the center of the Sun. Both the survival* $(P_{\alpha\alpha})$ *and transition* $(P_{\alpha\beta})$ *probabilities for different neutrino flavors are shown. In the energy regime of interest for weak-scale DM, the oscillations and absorptions are equally relevant. High-energy neutrinos are absorbed significantly by solar matter.*

where $E_{\rm max}/E = 4m_i M/(M + m_i)^2$ and $E_{\rm min}/E = v^2/(v^2 + v_{\circledast\rm esc}^2)$ are the maximal and minimal energy loss $\Delta$ that a particle can suffer in the scattering process, provided that it is captured. The term $|F_i(\Delta)|^2 = e^{-\Delta/E_0}$, where $E_0 = 3/2m_i r_i^2$ (for spin-dependent interactions) or $E_0 = 5/2m_i r_i^2$ (for spin-independent interactions)[35], with $r_i$ the effective nuclear radius, acts as a form factor capturing the suppression of the DM/nucleus cross section at large energy transfers.

The fraction of scatterings that lead to capture is largest for nuclei with mass $m_i$ comparable to the DM mass $M$, in which case $E_{\rm max}$ is maximized, and for DM particles that are slow (small $v$) and are traversing the central region of the body (large $v_{\circledast\rm esc}$), so that $E_{\rm min}$ is minimized. The latter observation is quite remarkable: standard direct detection searches are sensitive to high DM speeds (i.e., only DM particles with large enough kinetic energy deposit observable amounts of energy during scattering), while searches for neutrinos from the Sun are sensitive to low DM speeds. This distinction offers a potential avenue to break the degeneracy between the cross section and the speed distribution.

Fig. 6.10 (left) shows the total DM capture rate in the Sun for both spin-independent and spin-dependent scatterings of DM on nuclei. For concreteness, we set both the SI and SD cross section for scattering on a proton to $\sigma = 10^{-40}$ cm$^2$. The total capture rate is the sum of contributions from scattering on different nuclei, each proportional to its respective $\sigma_i$. For SI interactions, several elements in the solar composition contribute in similar amounts, either because they are very abundant (He) or

---

[35]See Ellis, Olive, Savage and Spanos (2010) [424] and references therein for the forms of these factors.

because they are heavy (Fe, O, Si, etc). In contrast, for the SD case, the capture rate is dominated by hydrogen, with nitrogen playing a minor role.

For heavy DM, $M \gg m_i \sim 100\,\text{GeV}$, DM can be captured only if it is very slow, $v < 2v_{\circledast\text{esc}}\sqrt{m_i/M}$. Consequently, in this limit the capture rate is proportional to $1/M^2$,

$$\Gamma_{\text{capt}} \approx \frac{5.9\ 10^{22}}{\text{sec}} \left(\frac{\rho_{\text{DM}}}{0.3\,\text{GeV/cm}^3}\right) \left(\frac{100\,\text{GeV}}{M}\right)^2 \left(\frac{270\,\text{km/sec}}{v_0}\right)^3 \frac{\sigma_{\text{SD}} + 1200\ \sigma_{\text{SI}}}{10^{-40}\,\text{cm}^2}, \qquad (6.65)$$

having approximated DM interactions with nuclei as a spin-dependent and a spin-independent cross section on nucleons (section 5.1).

As an aside, it is worth noticing that the SI capture rate is about 3 orders of magnitude larger than the SD one (assuming equal scattering cross sections, $\sigma_{\text{SI}} = \sigma_{\text{SD}}$). This is evident both in figure 6.10 (left) and in the approximate formula in eq. (6.65). Consequently, the expected neutrino signal for the SI case is larger by the same factor, and the bounds on SI scattering cross section about 3 orders of magnitude stronger. This is consistent with the results presented in fig.s 5.5 and 5.6. These figures also show that, as discussed in section 5.6.8, the bounds from nuclear recoil direct detection searches are very stringent for SI scattering, so that the solar neutrino bounds are competitive only for the SD case.

## 6.9.2 Computation of the neutrino energy spectra

The neutrino energy spectra from DM annihilations in the Sun, $dN_\nu/dE_\nu$ in eq. (6.55), are computed similarly to those in vacuum with two important differences.

First, one needs to take into account that DM annihilations occur within the dense matter of the Sun. Such an environment has a number of important consequences for the generated neutrino spectra. The hadrons produced in the annihilations shower and loose energy before they decay. Charmed and $b$-flavored hadrons are slowed down (to a different extent, $b$-hadrons are slowed more) and their energy distribution is reduced to a rapidly decreasing exponential as they move through solar matter. The neutrinos produced by the subsequent decays are therefore significantly less energetic than those produced by the same processes in vacuum. Positively charged pions and kaons lose significant energy and decay essentially at rest, producing monochromatic (in case of 2-body decays) or broader but still sharp (in case of 3-body decays) neutrino peaks. Neutrons and negatively charged pions are absorbed. The former create deuterons, the latter annihilate in nuclear matter. Negative kaons are captured. In these cases, very few or no neutrinos are produced. The leptons, which are produced in the annihilation showers, are also affected significantly. Negative muons are mostly captured. Positive muons are stopped and decay at rest. Taus, which have a shorter lifetime than muons, lose some energy before decaying. In addition, the struck nuclear matter can also produce unstable particles that create fluxes of secondary neutrinos. These complex processes can be modeled using dedicated MonteCarlo tools.

In summary, particles with a large cross section and/or long lifetimes tend to dissipate energy or even stop before decaying into neutrinos. Their decays give rise to large neutrino fluxes at low energies, typically below the detection threshold of large neutrino telescopes such as ICECUBE, but possibly within the sensitivity of SUPERKAMIOKANDE [461]. The high energy portion of the spectra, although reduced and smoothed out relative to annihilations in vacuum, however, remains to be prominent. Figure 6.11 (left) shows an example of the resulting neutrino fluxes.

Second, one needs to take into account the series of phenomena that neutrinos undergo as they propagate from their point of origin at the Sun's core to their eventual detection on Earth. These include: i) flavor oscillations within the Sun's matter, ii) interactions with the Sun's matter, iii) vacuum oscillations from the Sun to the Earth, and iv) oscillations within the Earth's matter, depending on the specific Earth segment traversed. The interactions in the Sun include two types of processes. First,

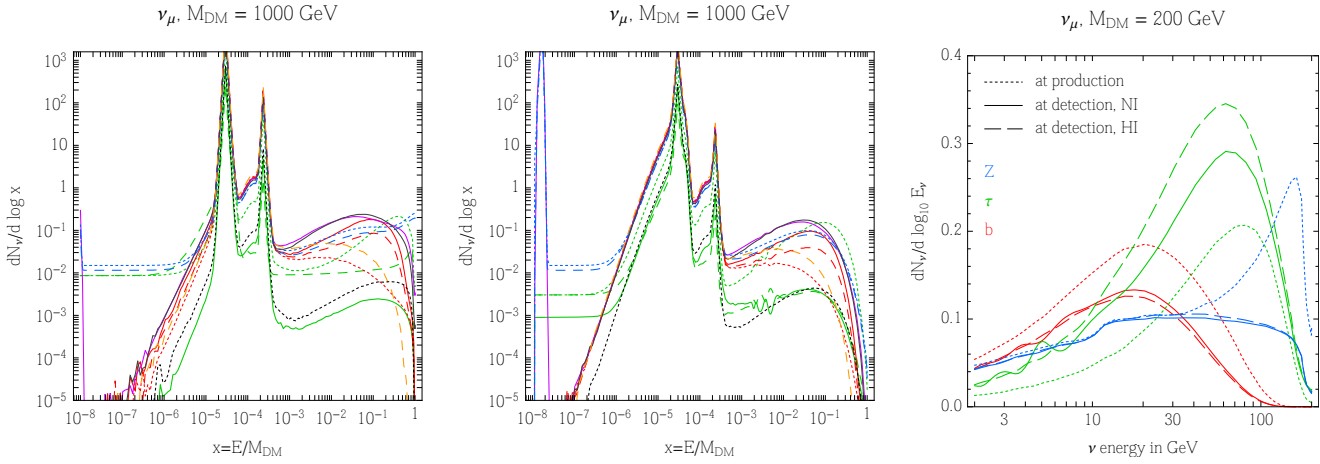

Figure 6.11: **Spectra of neutrinos from DM captured in the Sun**. *The color coding and style in the left and middle panels are the same as in figure 6.3. Left: $\nu_\mu$ spectra at production. The low energy peaks from the decays of stopped hadrons and leptons are clearly visible. Middle: $\nu_\mu$ spectra at detection, after propagation (interactions and oscillations, in solar matter, in vacuum and in terrestrial matter, assuming for definiteness normal hierarchy and neutrinos crossing vertically the Earth). Right: Direct comparisons of the $\nu_\mu$ spectra at production and detection, for a lighter DM particle and focusing on the energy range of interest for current neutrino telescopes. The considered annihilation channels are $ZZ$, $\tau^+\tau^-$ and $b\bar{b}$, and we show the results for both the normal and inverted neutrino hierarchy. (Figures from Baratella et al. (2014) in [424]).*

there are neutral current interactions where neutrinos scatter inelastically off nuclei, leading to partial energy loss. Second, there are charged current interactions, during which an incident neutrino scatters off a nucleus, subsequently producing a charged lepton and scattered hadrons. These leptons then decay back into neutrinos and anti-neutrinos of the same and different flavors, albeit of a lower energy. In particular, when the initial neutrino is $\nu_\tau$ ($\bar{\nu}_\tau$), the produced $\tau^-$ ($\tau^+$) decays promptly before losing energy, giving rise to another $\nu_\tau$ and $\bar{\nu}_e, \bar{\nu}_\mu$ (respectively $\bar{\nu}_\tau$ and $\nu_e, \nu_\mu$): this is the *tau regeneration* phenomenon. Since the neutrino scattering cross sections increase with energy, high energy neutrinos are efficiently absorbed as a consequence of interactions with matter (see figure 6.10 right) and their energy redistributed. Consequently, neutrinos produced in the center of the Sun only exit with sub-TeV energies.

Neutrino oscillations in the vacuum and within Earth's matter are governed by specific oscillation parameters commonly referred to as $\Delta m^2_{\text{sun}}$, $\Delta m^2_{\text{atm}}$, $\theta_{12}$, $\theta_{23}$, $\theta_{13}$ [438], including the still uncertain neutrino mass hierarchy ('normal' or 'inverted'). Inside the Sun, both coherent flavor oscillations and coherence-breaking interactions affect neutrino propagation and are equally important, as illustrated in figure 6.10 (right). Hence, a suitable formalism that includes both of these effects is required for a correct description. A mostly analytical method consists of studying the spatial evolution of the $3 \times 3$ matrix of neutrino densities, $\rho(E_\nu)$, and anti-neutrino densities, $\bar{\rho}(E_\nu)$. The diagonal entries of the density matrix represent the population of neutrinos of the corresponding flavor, whereas the off-diagonal entries quantify the quantum superposition of neutrino flavors. Using this formalism one finds that the fast oscillations average the off-diagonal elements to zero, without encoutering numerical issues. An alternative approach calculates probabilities using a fully numerical MonteCarlo simulation that follows the path of a single neutrino undergoing oscillations and interactions. The two approaches yield results that are in agreement

for all practical purposes.

In summary, propagation affects neutrino fluxes in three primary ways: it attenuates the fluxes, reshuffles neutrino flavors, and reduces neutrino energies. The middle panel in figure 6.11 shows an example of the resulting neutrino spectra after propagation, while the right panel shows the relative importance of propagation, albeit, for a lower value of $M$. The suppression of the high energy end of the neutrino spectrum is evident. The noticeable fluctuations around a few GeV can be attributed to the effects of flavor oscillations. Oscillations can also enhance the flux of a specific neutrino flavor, as showcased by the $\tau$ annihilation channel in the right panel in figure 6.11.

## 6.10   DM in stars

Several scenarios have been considered in which DM particles are captured inside stars (the Sun, white dwarfs or neutron stars) [462]. In this section we therefore generalize the discussion from the previous section, and summarize the main possibilities of capturing DM in stars and its phenomenology. See also Bramante and Raj in [1] for a dedicated review.

### 6.10.1   Other signals from DM capture in the Sun

Extending the discussion of section 6.9 beyond minimal models, one can consider the possibility that the DM particles captured in the Sun **annihilate into new hypothetical long-lived particles**. For example, *'secluded DM'* or *'dark sector'* models (see section 9.4) involve metastable mediators [463]. These mediators can escape from the dense solar core and subsequently decay into Standard Model particles. Depending on their lifetime, the decays can occur either in the outer layers of the Sun or in the empty space outside the Sun. In the former case, neutrinos remain the only particles that make it out of the Sun, however the signal is modified. In particular, the interactions of neutrinos with solar matter are less important so that neutrinos with higher energy can now escape, unlike in the scenarios involving DM annihilations in the solar core. If the mediators decay outside the Sun, these decays can result in gamma rays and charged particles, which can then be searched for. ANTARES, HAWC, FERMI and ICECUBE have put bounds on the neutrinos and gamma rays originating from these scenarios [463, 464].

The accretion of DM particles can result in an unconventional energy transport mechanism in the Sun's interior [465]. Because of their large mean free path, DM particles can efficiently transport energy radially (provided that they do not annihilate away too quickly), transferring heat from the deeper layers of the Sun to the external ones. This would **lower the temperature of the Sun's core**, leading to a decreased solar neutrino flux. Historically, in the 1980's, this mechanism was proposed as a solution to the solar neutrino problem (the deficit of neutrinos with respect to the predictions of the standard solar model) in the so-called *cosmion* model (WIMP-like particles with a mass of a few GeV and a now-excluded large cross-section $\sim 10^{-36}$ cm$^2$ for scattering on nuclei). The idea was abandoned when neutrino flavor oscillations were shown to be the explanation of the deficit.

The idea was revived around 2010 [466], to possibly explain the '*solar composition problem*' (or the '*solar abundance problem*'). This problem pertains to the disagreement between the determinations of the Sun's heavy element content from helioseismological data and the standard solar model predictions. One potential explanation centered on light asymmetric DM (see section 9.7), which by definition does not annihilate, and would therefore accumulate significantly in the solar core. The same line of reasoning was also used to derive constraints on WIMP-like DM. Subsequent careful studies showed, however, that the effect is too small to have any observable impact, at least for choices of parameters that are compatible with direct detection constraints. Nevertheless, for certain scenarios involving speed- or momentum-dependent DM scattering, or long-range interactions, the effects may still be significant.

## 6.10.2   DM capture in other stars

DM capture can become more significant for stars with characteristics that are more extreme than those of the Sun, or located in more extreme environments.

For **solar mass stars** [467], DM capture effects begin to be significant when these are immersed in a DM halo with density $\rho_{\rm DM} \gtrsim 10^3 - 10^5 \,{\rm GeV/cm}^3$ (depending on the effect), which is possibly realized close to the Galactic Center. There are two primary scenarios:

1. If DM particles annihilate, the consequences of DM capture depend on the amount of energy that DM annihilations inject into the stellar core, which in turn depends on the precise values of the environmental DM density, the DM speed distribution, the scattering cross section controlling the DM capture rate, and the mass and the composition of the star. Generally speaking, the energy released by DM annihilations extends the lifetimes of stars by counteracting the gravitational contraction and by contributing to the hydrostatic equilibrium balance. In optimal circumstances, a star could remain frozen in its evolution longer than the age of the Universe, in a condition that has been termed the **DM burner**. For red giant stars, the implications are similar but tend to be less pronounced. If DM is dissipative, the accumulation of DM is more efficient and the effects larger (see Peled and Volansky (2022)) [467]. However, given the complexity of stellar evolution, the detailed dependence on the parameters, and the difficulty of observing relatively normal stars very close to the GC, it is difficult to draw generic conclusions or bounds.

   Early very massive stars, referred to as **Population III stars** (see more extensive discussion in section 6.10.5 below), can also become DM burners by capturing massive quantities of DM, given that they formed in a dense DM environment at a high redshift [468]. If they burn DM, they can survive until the present epoch, and can be searched for as an anomalous stellar population: they would appear much larger and with a considerably lower surface temperature compared to normal stars with the same mass and metallicity. Due to the absence of reliable observations, and given the large dependence on the astrophysical parameters, no firm conclusions or bounds can be drawn on DM properties from this.

2. If DM particles do not annihilate, as is the case, for instance, for asymmetric DM (see section 9.7), they accumulate in the central region of the star. The accumulated DM can facilitate the transfer of energy towards the outer layers, similar to the scenario described for the Sun in section 6.10.1. While the effects on the Sun are mostly negligible, stars situated within dense DM environments can experience significant heat transport, leading to notable alterations in their evolutionary path. The net effect is somewhat opposite to the case of the DM burner; the star has to increase its core nuclear burning intensity to compensate for the reduction in temperature caused by the DM transport, therefore modifying its position in the main sequence of the Hertzsprung-Russel diagram, making itself recognizable (bigger and more luminous),[36] as worked out in particular by Iocco et al. (2012) [467]. However, no conclusive evidence of such peculiar stars has been presented to date.

## 6.10.3   DM capture in white dwarfs

White Dwarfs (WD) represent the final evolutionary stage for stars with mass $M_{\rm wd}$ below the Chandrasekhar limit[37] of $1.44 \, M_\circledast \sim M_{\rm Pl}^3/m_p^2$, meaning that these majority of stars lack the mass to collapse into either a neutron star or black hole. Instead, the gravitational collapse of the stellar core remnant is halted by the degeneracy pressure of non-relativistic electrons. As a result, the stellar mass ends

---

[36]In principle, this also alters the star's neutrino output, although such change we are unable to detect.

[37] Chandrasekhar originally derived this limit by modelling electrons as an ideal Fermi gas, and also derived its approximate dependence on fundamental constants, $M_{\rm Pl}$ and $m_p$.

up concentrated in a radius $R_{\rm wd} \sim M_{\rm Pl}/m_e m_p$, comparable to the Earth's radius.[38] White dwarfs are mostly composed of the last element which could be burnt by the progenitor star — this element varies with the progenitor's mass, being helium for lighter stars and either carbon or oxygen for the more massive ones. Survey data indicate that the white dwarf mass distribution is peaked at $0.6 M_\odot$. A white dwarf possesses a non-relativistic escape velocity, $v_{\rm esc} \sim \sqrt{2 m_e/m_p} \sim 0.03$. The rough estimate $\rho_{\rm wd} \sim 3 m_e^3 m_p/4\pi \sim$ few $10^6 \, {\rm g/cm}^3$ for the white dwarf density shows that it is parametrically higher than the density of ordinary matter, estimated as approximately $\rho_{\rm ord} \sim \alpha^3 \, m_e^3 m_p \sim$ few ${\rm g/cm}^3$, where $\alpha$ is the fine structure constant.

The computation of DM capture in white dwarfs proceeds similarly to that in the Sun or ordinary stars, with the advantage that the higher escape velocity in white dwarfs compared to stars enhances the DM capture rate, see eq. (6.62) or (6.63). One needs to take into account the peculiar core composition of white dwarfs, and DM capture due to scattering on the degenerate electron sea can also be considered. As discussed in section 6.10.2, DM capture in such bodies can lead to two distinct observable consequences:

1. **If DM particles annihilate,** the impact of energy injection[39] is more significant in white dwarfs compared to stars, owing to the lower temperatures of white dwarfs. The WD can become a **DM burner** [469]: it possesses an internal source of energy despite the depleted nuclear fuel. Consequently, its surface temperature can rise to $1.4 \, 10^5 \, {\rm K}$, which is at the upper end of the normal range: typical white dwarfs have surface temperatures in the range $(0.8-4) \, 10^4 \, {\rm K}$. Observations of old, but still relatively hot WDs, could thus provide a valuable test of these scenarios. In turn, the observations of WDs that are not anomalously luminous can constraint DM properties. Dasgupta et al. (2019) and Bell et al. (2021) [469] used the observations of WDs in the globular cluster M4 to derive bounds on the spin-independent DM-nucleon cross section, as well as on the SI DM-electron scattering cross section. The respective bounds are of the order of $\sigma_{\rm SI} < 10^{-(42-44)} \, {\rm cm}^2$ (for $100 \, {\rm keV} \lesssim M \lesssim 100 \, {\rm TeV}$) and $\sigma_{\rm SI}^{\rm el} < 10^{-41} \, {\rm cm}^2$ (for $M \gtrsim 100 \, {\rm MeV}$). The bounds on $\sigma_{\rm SI}$ are less stringent than the ones from direct detection experiments, for electroweak scale DM masses, see fig. 5.5 (bottom). However, the WD bounds extend to much lighter DM masses (for which, however, evaporation needs to be carefully taken into account), and are also more stringent for heavier DM. In contrast, the WD constraints on DM-electron scattering are more stringent than direct detection bounds over the whole applicable DM mass range, see fig. 5.12. Nevertheless, the WD limits warrant caution due to ongoing discussions on the DM composition in globular clusters, as briefly touched upon in section 6.2. Since WDs are assumed to be composed solely of one of the following elements, $^4$He, $^{12}$C or $^{16}$O, all of which have spin-less nuclei, no bounds on SD scattering can be placed using these observations.

2. **If DM annihilations are negligible**, DM accumulates in the core of the white dwarf. Even a small amount of DM (so that the total mass of WD-captured DM, $M_{\rm DM}$, satisfies $M_{\rm DM} \ll M_{\rm wd}$) can trigger a **Type-Ia supernova explosion** of WD [470]. Since evaporation is not relevant for large DM masses (see the discussion in page 224 for the case of the Sun), the captured DM particles rapidly fall to the centre of WD, forming a thermal sphere with radius (see eq. (6.56))

$$ r_{\rm DM} \sim \left( \frac{3 \, T_{\rm wd}}{2\pi \, G \rho_{\rm wd} M} \right)^{1/2} \sim 300 \, {\rm m} \qquad \text{for } T_{\rm wd} \sim 10^7 \, {\rm K} \text{ and } M \sim 10^6 \, {\rm GeV}, \qquad (6.66) $$

---

[38]This radius can be estimated by minimising $U_{\rm gravity} \sim -M/M_{\rm Pl}^2 R$ plus $U_{\rm quantum} \sim Q^{5/3}/m_e R^2$ where $Q \sim (M_{\rm Pl}/m_p)^3$ is the Chandrasekhar bound on the number $Q$ of electrons and protons.

[39]The case in which annihilations occur into relatively long-lived mediators that decay outside of the star has been addressed by Acevedo, Leane and Santos-Olmsted (2023) [469]. In this case, which is equivalent to the one discussed in section 6.10.1 for the Sun, no energy is injected in the star. Instead, the decays of the mediators produce $\gamma$-rays that can be searched for.

where $T_{\mathrm{wd}}$ is a typical core temperature and $\rho_{\mathrm{wd}} \sim 10^6$ g/cm$^3$ a typical core density. Notice that 'normal' stars have a much lower core density and also a somewhat higher core temperature. This implies that in white dwarfs DM capture (eq. (6.59)) is more efficient and the DM sphere more compact. This is why white dwarfs, and compact objects in general, can give stronger constraints on DM than ordinary stars.

If the DM density $M_{\mathrm{DM}}/r_{\mathrm{DM}}^3$ exceeds the matter density of the host white dwarf, then DM starts self-gravitating and collapses, possibly forming a small black hole. The thermal energy released during the collapse can trigger nuclear fusion and therefore ignite a supernova explosion. Alternatively, the black hole forms, and its subsequent evaporation triggers the supernova explosion.[40] Observations of specific white dwarfs, which remained stable and did not explode, combined with the recorded supernova rates, provide evidence against this scenario. The resulting bounds on the SI or SD DM-nucleon scattering cross section $\sigma_{\mathrm{SI,SD}}^N$ can be stronger than the bounds from direct detection [470]. They are relevant for $M \gtrsim 10^6$ GeV and can reach $\sigma_{\mathrm{SI,SD}}^N \sim 10^{-43}$ cm$^2$, as plotted in fig. 5.5.

## 6.10.4   DM capture in neutron stars

DM capture can be even more important for denser compact objects such as neutron stars (NS). These are remnants of massive super-Chandrasekhar stars, originally possessing a mass typically between 10 and 25 $M_{\circledast}$, which underwent a supernova explosion. Neutron stars are supported by neutron degeneracy pressure. Estimates analogous to white dwarfs in section 6.10.3 apply, with $m_e$ replaced by $m_n$. That is, neutron stars have a typical radius of $R_{\mathrm{ns}} \sim M_{\mathrm{Pl}}/m_n^2 \sim 10$ km, a mass around $M_{\mathrm{ns}} \sim 1.4 M_{\circledast}$ and density $\rho_{\mathrm{ns}} \sim m_n^4 \sim 10^{15}$ g/cm$^3$ close to that of nuclear matter. This predicted large $\rho_{\mathrm{ns}}$ density is observationally confirmed by the existence of pulsars with fast $T \sim$ ms period, as the centrifugal force would exceed the gravitational force if $T \lesssim M_{\mathrm{Pl}}/\sqrt{\rho_{\mathrm{ns}}}$: fast-spinning gravitational stable bound states must be dense. The total number of nucleons is $N_{\mathrm{ns}} \sim (M_{\mathrm{Pl}}/m_n)^3 \sim 2 \, 10^{50}$. Since there are no small parameters, the escape velocity $v_{\mathrm{esc}}^2 \approx 2G \, M_{\mathrm{ns}}/R_{\mathrm{ns}}$ is comparable to the speed of light. Gravity is so strong that DM particles attracted within the impact parameter ($v$ is the DM speed in the halo, far from the star)

$$\frac{b}{R_{\mathrm{ns}}} = \frac{v_{\mathrm{esc}}/v}{\sqrt{1 - v_{\mathrm{esc}}^2}}, \tag{6.67}$$

hit the neutron star with relativistic energy, $E \sim \gamma M$, where $\gamma \sim 1.3$. DM is accreted at a rate $dM_{\mathrm{DM}}/dt \sim \rho_{\mathrm{DM}} v \, \pi b^2 \min(1, \sigma_N/\sigma_*)$. Over a time interval $\Delta t$ this results in a total accumulated DM mass of

$$M_{\mathrm{DM}} \sim 10^{11} \, \mathrm{kg} \frac{\rho_{\mathrm{DM}}}{0.3 \, \mathrm{GeV}/\,\mathrm{cm}^3} \frac{\Delta t}{10^{10} \, \mathrm{yr}} \frac{b^2}{R_{\mathrm{ns}}^2} \min\left(1, \frac{\sigma_{\mathrm{SI,SD}}^n}{\sigma_*}\right). \tag{6.68}$$

Here, $\sigma_{\mathrm{SI,SD}}^n$ is the SI or SD DM/neutron cross section and $\sigma_*$ its critical value above which all DM that hits the neutron star gets geometrically captured

$$\sigma_* \sim \frac{\pi R_{\mathrm{ns}}^2}{N_{\mathrm{ns}}} \sim 10^{-44} \, \mathrm{cm}^2. \tag{6.69}$$

---

[40]If DM is bosonic and $T < T_{\mathrm{BEC}} \sim n^{2/3}/M$, then DM does not have a thermal Maxwellian distribution, but rather forms a Bose-Einstein condensate (BEC). The DM sphere has a smaller radius $R_{\mathrm{BEC}} \sim (GM^2\rho_{\mathrm{dw}})^{-1/4}$, and forms a black hole more easily. If the black hole is big enough to undergo the Bondi-Hoyle accretion until it engulfs the white dwarf, instead of evaporating through Hawking radiation, $\dot{M}_{\mathrm{BH}} \sim -M_{\mathrm{Pl}}^4/M_{\mathrm{BH}}^2$, this also leads to bounds on the DM parameter space, which are, however, rather complex (see, e.g., Janish et al. (2019) in [470]).

That is, DM is captured even for very small values of $\sigma_*$, comparable to bounds from direct detection searches.

The captured DM can lead to several potentially observable signatures, listed below. While the first two are extensions of those discussed for other astrophysical bodies in preceding subsections, the third is specific to neutron stars.

1. **If accumulated DM particles *annihilate*,** the neutron star achieves the condition of **DM burner**, and its surface temperature increases [471]. Neutron stars form at large temperature $T \sim m_n$ and then undergo cooling, initially due to neutrino emissions and later because of photon emissions. In the absence of heating processes, temperatures below 1000 K are reached after about 10 Myr. Detecting old neutron stars with surface temperatures somewhat higher than this would allow to probe this scenario.

2. **If accumulated DM particles *do not annihilate*** [472], they form a thermal sphere at the center, with a radius $r_{\mathrm{DM}} \sim 30$ cm for $T_{\mathrm{ns}} \sim 10^5$ K and $M \sim 100$ GeV. **A black hole eventually forms** and consumes the neutron star. The mere existence of neutron stars in various DM backgrounds (i.e., the fact that they were not destroyed by DM), allows to place stringent constraints on the DM/neutron scattering cross section. The detailed bounds (see in [472]) depend on many parameters, including whether or not a Bose-Einstein condensate of DM forms, similar to the situation in white dwarfs. The bounds derived by Bramante et al. (2018) [472] reach $\sigma_{\mathrm{SI,SD}}^N \sim 10^{-47}$ cm$^2$ for $M \approx 10^{10}$ GeV, as plotted in fig. 5.5. Garani et al. (2018) find even stronger constraints in some specific configurations.

3. Regardless of whether they eventually annihilate or not, captured DM particles get accelerated to relativistic speeds, fall onto the neutron star and deposit their kinetic energy. This **DM *kinetic* heating of neutron stars** [473] keeps the surface at a temperature $T_{\mathrm{surface}} \approx (\rho_{\mathrm{DM}}/v_{\mathrm{DM}})^{1/4}$ (about 1700 K for neutron stars in our vicinity), for which the heating rate $dE_{\mathrm{DM}}/dt$ is in equilibrium with the photon emission rate, $W = 4\pi R^2 \pi^2 T_{\mathrm{surface}}^4/60$. This happens if $\sigma_{\mathrm{SI,SD}}^n \gtrsim \sigma_*$ and GeV $\lesssim M \lesssim$ PeV. The mass range can be understood as follows. Upon single scattering, a falling DM transfers to neutrons an energy

$$\Delta E = \frac{m_n \gamma^2 v_{\mathrm{esc}}^2}{1 + m_n^2/M^2 + 2\gamma m_n/M} \langle 1 - \cos\theta_{\mathrm{CM}} \rangle, \tag{6.70}$$

which is larger than the redshifted kinetic energy of the free DM, $\gamma M v_{\mathrm{esc}}^2/2$, if $M \lesssim 10^6$ GeV. For larger $M$ multiple scatterings are needed to capture DM, and as a result $\sigma_*$ increases as $M/10^6$ GeV for heavier DM. On the other hand, for $M \lesssim m_n$ the transferred energy $\Delta E$ is so low that due to Pauli blocking (in a neutron star the neutrons occupy energy levels up to the Fermi momentum of about $m_n$) $\sigma_*$ increases as $m_n/M$, if DM mass is lowered.

4. **MeV to GeV DM** that interacts strongly could detectably alter the structure of neutron stars [474].

Next, we discuss the observational status of items 1 and 3. Presently, meaningful tests are still lacking, since the observational sensitivities to the temperature of neutron stars tend to be about a factor of 20-100 higher than the expected effects due to DM. The James Webb Space Telescope could observe $T \sim 2000$ K for certain nearby neutron stars.

Furthermore, there exist various sources of backgrounds. Neutron stars could be heated by other mechanisms, such as the conversion of their rotational energy trough magnetic fields, or due to the accretion of normal matter onto their surface (resulting in pronounced heating of the magnetic poles), crust cracking, vortex friction, etc. It is therefore essential to focus on isolated neutron stars situated in regions where DM density surpasses that of normal matter. In the interstellar medium, DM and matter

densities are comparable, which could potentially obscure any additional heating effects due to DM. Fortunately, $X$-ray data suggest that since about 10 Myr the Sun has resided in a 'local bubble' where the matter density is nearly a factor of 100 less than the average. In this environment, DM annihilations would heat neutron stars up to $\sim 2500$ K.

Should future telescopes be able to test neutron star heating due to DM, the DM/nucleon cross sections will be probed down to $\sigma_{\rm SI,SD}^{N} \lesssim 10^{-46}$ cm$^2$ for 1 GeV $\lesssim M \lesssim 10^6$ GeV [471, 473]. This surpasses current spin-dependent direct detection bounds, while it is not competitive with the spin-independent bounds. Nevertheless, such findings would still be interesting, since, as discussed in section 5, various effective operators induce small direct detection cross sections, suppressed by the DM velocity or momentum transferred. These suppressions are lifted in relativistic collisions of DM on neutron stars. For non-minimal models, such as inelastic DM (section 5.5.1), relativistic DM collisions on neutron stars would probe much higher DM inelasticities up to $\sim m_n$ (covering, for example, the mass splittings predicted by theories where DM is part of a single electroweak multiplet, see section 9.3.4).

## 6.10.5   Dark stars?

The role of DM in the formation and evolution of early stars has also been considered in depth. The first stars in the Universe, termed Population III stars, formed out of diffuse gas clouds situated within and concentric to large DM halos (typical mass of $\sim 10^6 \, M_\odot$, and composed of approximately $\sim 85\%$ DM, consistent with the Universe's general composition). In the standard picture, both the DM halo and the gas cloud contract, and eventually the baryonic matter ignites nuclear fusion at the center of the cloud, which then hydrostatically supports the newly formed star, halting the collapse.

Beginning in 2007, a series of works speculated that, instead, the DM halo concentration could ignite DM annihilations at its center. This would lead to a halt in the collapse of the gas cloud prior to the onset of nuclear fusion, and to the formation of a pseudo-stellar object, a '*dark star*' [475], composed of hydrogen and helium, and sustained by DM annihilations instead of nuclear fusion. For this mechanism to work, a sufficiently high DM density is essential. Notably, in the early Universe, DM density was higher at redshift $z$ by a factor $(1 + z)^3$. Moreover, the first stars form exactly in the centers of their host DM halos, where the density is highest. The process is sustained if the heat produced by DM is efficiently trapped inside the star (which is the case because of the high baryonic density, akin to the Sun's core) so that the heating rate can be higher than the baryonic cooling mechanisms (which is found to occur when the gas density exceeds $10^{13}$ particles/cm$^3$ for a typical WIMP-like DM). Note also the conceptual difference with the 'DM burner' case discussed earlier: in that case, DM particles accumulate in the center of early stars by scattering on the stellar matter, leading to DM capture, while in dark stars DM gets trapped during a concentric collapse. For DM capture, DM particles need to possess a significant scattering cross section with baryons, while dark stars can in principle be born even for mostly gravitationally-interacting DM, provided that it then annihilates into SM particles. The canonical thermal relic cross section $\sim 2\,10^{-26}$ cm$^3$/s in eq. (4.13) is sufficient for the mechanism to work.

This unconventional stellar phase would start very early, at redshift $z \sim 10 - 50$, and could last for millions to billions of years, contingent on the availability of DM fuel. If would create anomalous stars with very heavy masses approaching $\sim 1000 \, M_\oplus$, large sizes of up to $\sim 200 R_\oplus$, high luminosities up to $10^6 \, L_\oplus$ and a low surface temperature below $10^4$ K. These characteristics would make them observationally very distinct from standard Population III stars, and potentially detectable at very high redshifts with the James Webb Space Telescope. Dark stars can also alter the reionization history of the Universe (negatively, their low surface temperature implies that they would emit less ionizing radiation compared to normal Population III stars) and modify the cosmic microwave background [475].

Upon depletion of DM fuel, dark stars would transition into heavy main-sequence stars. The most massive among them would eventually collapse into anomalously massive black holes, possibly providing a seed to the formation of early supermassive black holes.

Certain simulations [475] claimed that dark stars do not form, although this could be due to their limited resolution. Observationally, no evidence of dark stars has been found so far, and no general bounds have been derived, though C. Ilie et al. (2023) [475] proposed candidate objects consistent with dark stars.

### 6.10.6  DM capture in planets

Some of the physical concepts discussed above can be applied also to the capture of DM by planets (the Earth, other solar system planets, or exoplanets), leading to noteworthy constraints [188].

1. **If DM particles can self-annihilate**, the energy deposition rate from the annihilation products should not exceed the measured ***heat flow*** emanating from the Earth's interior. This was first noted by Mack et al. (2007) and then refined and extended in Bramante et al. (2020). The constraints apply to DM masses 100 MeV $\lesssim M \lesssim 10^{10}$ GeV and span the very broad range $10^{-38}$ cm$^2 \lesssim \sigma_{\text{SI}} \lesssim 10^{-16}$ cm$^2$ (for spin-independent scattering, see figure 5.5) and $10^{-33}$ cm$^2 \lesssim \sigma_{\text{SD}} \lesssim 10^{-11}$ cm$^2$ (for spin-dependent scattering).

   Some pioneering studies [188] also considered the heat flow from inside Jupiter, Saturn, Uranus and Pluto, but this did not yield particularly relevant constraints. Similarly, the requirement that the DM heat injections did not halt the contraction of the proto-planetary gas cloud and thus the formation of Jupiter (or Jovian planets) allows to impose bounds, which are however not particularly competitive.

   Recent investigations have also examined Mars and exoplanets. In the context of exoplanets, assuming that accurate temperature measurement will be possible (e.g., in the infrared with the James Webb Space Telescope), the sensitivity on the spin-dependent cross section on protons could reach $\sigma_{\text{SD}}^p \lesssim$ few $10^{-37}$ cm$^2$ for GeV and sub-GeV DM. This improved sensitivity compared to Earth observations can be attributed to the expected lower temperature of exoplanets, and to their large number.

2. **If DM particles do not annihilate,** they accumulate at the center of the planet and form a black hole, which again leads to constraints. For example, Acevedo et al. (2021) [188] determined that DM with mass $10^7$ GeV $\lesssim M \lesssim 10^{10}$ GeV and spin-independent cross-section $10^{-32}$ cm$^2 \lesssim \sigma_{\text{SI}} \lesssim 10^{-17}$ cm$^2$ is ruled out, otherwise the BH would have destroyed the Earth within a billion years. The updated analysis in Ray (2023) [188] finds that celestial objects with large sizes and relatively low core temperatures, such as Jupiter, are more suitable and can exclude a slightly larger area of the parameter space. For even larger masses, and proportionally larger cross-sections, the BH would evaporate and overheat the planet.

## 6.11  DM in the dark ages: CMB, reionization and heating

Dark ages are the epoch between recombination at $z \approx 1100$ and the formation of the first stars at $z \gtrsim 10$. Annihilations or decays of DM particles during the cosmic dark ages inject energy into the medium, which causes heating and additional ionization of the ambient hydrogen gas. The resulting free electrons scatter on the CMB photons and modify the measured properties of the CMB, providing a mechanism to probe the DM annihilation cross section or decay rate. The ensuing constraints are quite competitive with many of the other indirect detection bounds coming from the Galaxy or the local universe (see figures 6.14, 6.15 and 6.16), especially for light DM. They are also quite robust, i.e., subject to a lesser extent to astrophysical uncertainties such as the DM distribution, the galactic propagation

of cosmic rays and the environmental properties (galactic magnetic fields, ambient light distribution...). They do, however, remain sensitive to the velocity dependence of the annihilation cross-section, as we discuss below.

The fraction $x_{\rm ion}(z)$ of singly ionized atoms, which would normally hover around $10^{-4}$ for $z \lesssim$ few $10^2$ in the absence of exotic ionization processes, is affected by DM annihilations or decays. It follows the basic evolution equation

$$n_{\rm A}(1+z)^3 \frac{dx_{\rm ion}(z)}{dt} = I(z) - R(z). \tag{6.71}$$

On the right-hand side, we have the rate of ionization per volume $I(z)$, which tends to increase $x_{\rm ion}$, and the recombination rate per volume, $R(z) = R_{\rm H}(z) + R_{\rm He}(z)$, with which the hydrogen and helium atoms in the IGM recombine while ionization is ongoing. We omit the explicit expressions for these recombination rates here for brevity, but they can be found in the literature [476]. The rate of ionizations per volume due to DM, at a given redshift $z$, is given by

$$I(z) = \frac{\eta_{\rm ion}}{e_{\rm ion}} \frac{dE}{dt}(z), \tag{6.72}$$

where $e_{\rm ion}$ is the average ionization energy per baryon (e.g., 13.6 eV for hydrogen). The factor $\eta_{\rm ion}$ is the fraction of the energy that goes into ionizations, while the remainder contributes to heating and atomic excitations. The value of this fraction depends on $x_{\rm ion}(z)$ itself, and can in first approximation be taken to be (see, e.g., Chen and Kamionkowski (2004) [476] and references therein)

$$\eta_{\rm ion} = \frac{1 - x_{\rm ion}(z)}{3}. \tag{6.73}$$

The last term in eq. (6.72) represents the energy injection rate from dark matter per unit volume. It is given by

$$\frac{dE}{dt}(z) = \begin{cases} \rho^2(1+z)^6 f(z) \dfrac{\langle \sigma v \rangle}{M} & \text{(DM annihilation)}, \\[2mm] \rho(1+z)^3 f(z) \, \Gamma & \text{(DM decay)}. \end{cases} \tag{6.74}$$

Its form can be easily understood. Focusing on annihilating DM, the energy injection rate is determined by the number density of annihilating DM particles, $n = \rho/M$, squared, and rescaled to redshift $z$ by the factor $(1+z)^3$, multiplied by the probability of annihilations (given by the cross section $\langle \sigma v \rangle$), and by the energy injected in each annihilation (equal to $M$). One power of $M$ cancels in the final expression, giving a rate proportional to $\langle \sigma v \rangle / M$. For decaying DM, the energy injection rate is proportional to the decay rate $\Gamma$, and a single power of $n$, so that the dependence on $M$ completely cancels out. The factor $f(z)$ expresses the fraction of the total DM energy which is absorbed by the gas. It incorporates all the complexity of *how* the DM annihilation/decay products interact with the gas and *when* they do so. Distinct values of $f(z)$ can be calculated for various annihilation/decay species (e.g., $e^\pm$, hadrons,..., excluding neutrinos, which do not interact electromagnetically and therefore do not deposit any energy), for different spectra (the $dN/dE$ in eq. (6.1)), for different injection redshifts $z$, and taking into account that the absorption in general occurs at a later redshift $z'$, after some cooling, etc. Their values can be found tabulated in the literature (in particular Slatyer (2016) in [477]).

In practice, it turns out that the impact of annihilating DM is well characterized by a single parameter: the excess ionization at redshift $z \sim 600$. For DM decay, the signal is similarly dominated by redshift $z \sim 300$. It is thus useful to introduce an effective redshift-independent factor $f_{\rm eff}$. Note that this still depends on the specific DM particle physics model in a non-trivial way. The impact of DM annihilations can thereby be encoded in the factor $p_{\rm ann} = f_{\rm eff} \langle \sigma v \rangle / M$, and in $p_{\rm dec} = f_{\rm eff} \, \Gamma$ for DM decays.

Based on these physical principles, the evolution equation (6.71) can be solved for $x_{\rm ion}$ at any redshift

$z > 6$.[41] This can be done at varying levels of refinement, either semi-analytically or using dedicated reionization codes such as `Recfast`, `HyRec` or `CosmoRec`.

### 6.11.1 Optical depth

An initial series of studies determined DM constraints using a rather simplistic criterion that DM-induced free electrons should not alter the **optical depth**, experienced by CMB photons during their journey from the last scattering surface to us, beyond the measurements reported by CMB surveys like WMAP or the more recent PLANCK satellites [476]. The optical depth $\tau_{\rm op}$ is given by the integral (over the travel time of the photon) of the number density of free electrons $n_e(z)$, i.e., the scattering targets, multiplied by the Thomson scattering cross section $\sigma_{\rm T} = 8\pi r_e^2/3 = 0.6652$ barn (with $r_e$ the electron classical radius)

$$\tau_{\rm op} = -\int n_e(z)\,\sigma_{\rm T}\,\frac{dt}{dz}, \tag{6.75}$$

where $dt/dz$ is the standard $\Lambda$CDM relation between time and redshift in eq. (C.13). In addition, $n_e(z) = n_{\rm A}(1+z)^3\,x_{\rm ion}(z)$, where $n_{\rm A}$ is the number density of atoms today. The bounds obtained in this way constrain, for instance, $\langle\sigma v\rangle \lesssim 10^{-24}$ cm$^3$/s for $M \approx 100\,{\rm GeV}$.

### 6.11.2 CMB anisotropies

A subsequent set of studies significantly improved the analysis by examining the effects of DM annihilations and decays on the **CMB anisotropies** [477]. Rather than relying on the optical depth as a proxy, these works used detailed CMB codes such as `Camb` or `Class` to compute the modified power spectrum in presence of an increased ionization fraction, and compared with the refined PLANCK data. The resulting bounds are discussed in section 6.13.2 and have gained significant traction in the literature. Note that the bounds on $\langle\sigma v\rangle$ scale linearly with $M$, while those on the DM decay lifetime $\tau$. These bounds are more stringent than those based solely on optical depth. For instance, they constrain $\langle\sigma v\rangle \lesssim 10^{-25}$ cm$^3$/s for $M \approx 100$ GeV.

As mentioned above, the CMB bounds are robust with respect to many astrophysical uncertainties. Specifically, since the signal is dominated by high redshifts, prior to the onset of structure formation, the bounds do not depend on the details of DM distribution; the density $\rho$ in eq. (6.74) is from the smooth DM background, which is a well known quantity in the standard cosmology. That is, there is no need to specify the halo formation history, the halo profiles, etc. However, for annihilating DM the CMB bounds do exhibit sensitivity to the velocity dependence of the annihilation cross section. For $p$-wave DM annihilation, i.e., $\langle\sigma v\rangle \propto v^2$, these bounds weaken considerably, because the energy injection from DM gets suppressed due to DM being very cold (slow) during the dark ages. Put differently, a large present day annihilation cross section is still allowed, because it was suppressed during the dark ages.

### 6.11.3 Temperature of the intergalactic medium

A (smaller) series of studies explored the effect of DM energy injection on the **temperature of the intergalactic medium** $T_{\rm igm}$ [478]. Existing measurements of $T_{\rm igm}$ place direct constraints on the maximal allowed DM-induced heating. Moreover, if the temperature of the IGM is too high, gas clouds would not collapse and star formation would be delayed or impeded, allowing to impose indirect constraints on DM.

---

[41]Below redshift 6 all the hydrogen and helium gas is assumed to be ionized, while helium is also doubly ionized below redshift $z = 3$, as suggested by the Gunn-Peterson test: the absence of absorption at Lyman-$\alpha$ frequency in the spectra of quasars located at $z < 6$. Hence $x_{\rm ion}(z < 6) \approx 1$.

The temperature $T_{\rm igm}$ follows a basic evolution equation somewhat analogous to eq. (6.71),

$$n_{\rm A}(1+z)^3 \frac{dT_{\rm igm}(z)}{dt} = -X(z) + C(z) + W(z). \qquad (6.76)$$

The first term on the r.h.s. describes the usual (adiabatic) cooling of the gas due to the expansion of the Universe. It reads $X(z) = 2n_{\rm A}H(z)(1+z)^{3/2}\,T_{\rm igm}(z)$ and would lead to $T_{\rm igm}(z) \propto (1+z)^2$. The second term, $C(z) \propto T_{\rm CMB}(z) - T_{\rm igm}(z)$, accounts for the coupling between the inter-galactic gas and the CMB photons, that have a redshift-dependent temperature $T_{\rm CMB}$. When the inter-galactic gas is hotter than the surrounding CMB, some of its energy is transferred to the photons and therefore the gas 'Compton-cools'. Conversely, if the gas is colder than the CMB, it heats up. The $C(z)$ term is only sketched here: we refer to the relevant literature for its precise form and the details concerning its derivation.[42] The third term on the right side of eq. (6.76) encodes the warming of the inter-galactic gas due to DM energy release, and can be expressed as

$$W(z) = \frac{2}{3}\eta_{\rm heat}(z)\frac{dE}{dt}(z), \qquad (6.77)$$

where, similarly to eq. (6.73), the factor

$$\eta_{\rm heat}\big(x_{\rm ion}(z)\big) = K \left[1 - (1 - x_{\rm ion}^a)^b\right], \qquad (6.78)$$

expresses the fact that only a fraction of the DM injected energy goes toward heating. Here, $K = 0.9971$, $a = 0.2663$, and $b = 1.3163$ are the values estimated in the literature.

Solving the coupled[43] differential equations (6.71) and (6.76), with appropriate expressions for the $f(z)$ factors relevant for heating, allows to obtain a prediction for $T_{\rm igm}(z)$ in the presence of DM as an exotic source of heating. Unlike in the CMB case, however, the potentially observable modifications to the temperature history depend primarily on physics at much lower redshifts, and therefore are very sensitive to structure formation history. The density $\rho$ in eq. (6.74) has to be multiplied by a boost factor $B(z)$ which takes into account the DM collapse into gravitationally bound structures, their abundance, their density profile, etc. The resulting bounds on the DM annihilation or decay rates are weaker than those derived from the optical depth constraint, and therefore also less competitive than those from the CMB. For instance, they constrain $\langle\sigma v\rangle \lesssim 2\,10^{-23}$ cm$^3$/s for $M \approx 100$ GeV, for DM annihilations into $\mu^+\mu^-$ and assuming an NFW profile for the DM halos.

## 6.12  DM and Big Bang Nucleosynthesis

Historically, BBN has been pivotal in providing evidence for Dark Matter (DM), as discussed in section 1.3.5. Subsequently, BBN has also been employed to probe DM properties and establish constraints, which we briefly overview in this section [479].

To begin with, let us consider the case of **weak-scale Dark Matter**. The injection of particles from DM annihilations or decays during or immediately after BBN can lead to the destruction of some species of the newly formed light nuclei and to the overproduction of others, thereby altering their final abundances, and potentially spoiling the agreement with observations. This process is conceptually similar to DM energy injection during the dark ages, discussed in section 6.11. The differential equations to be solved

---

[42]The same physics is relevant for the 21 cm measurements, in which an anomaly has been detected by EDGES, see section 8.4.5.

[43]The equations are coupled since $T_{\rm igm}$ enters the expressions for the recombination rates $R(z)$. The recombination codes mentioned above also allow to calculate the modified temperature history. The `DarkHistory` code does an even more refined job.

here, the analogues of eq.s (6.71) and (6.76) in section 6.11, are those governing the relative abundances $Y_i \equiv n_i/n_b$ of each light element $i = (p, n, \text{D}, {}^4\text{He}, {}^3\text{He}, {}^7\text{Li} \ldots)$, normalized to the total number density of baryons $n_b$. Schematically, they can be written as

$$\frac{dY_i}{dt} = -\Gamma_i Y_i + \sum \left( \Gamma_{ji} Y_j + \Gamma_{kji} Y_k Y_j + \ldots \right), \tag{6.79}$$

where $\Gamma_i$ is the decay rate of species $i$, $\Gamma_{ji}$ is the rate of the $i$-producing decay $j \to i + X$ (with $X$ denoting any other particles), $\Gamma_{kji}$ is the rate of the process $k + j \to i + X$, etc. These interconversion rates depend on the temperature and on the environment. The complex network of such equations is typically treated with dedicated codes such as `Primat`, `Parthenope`, `AlterBBN` or earlier ones [480], which solve up to several tens of coupled equations for as many nuclear species. DM annihilations or decays perturb the system in a complex way, by modifying the environment and adding an exotic source of coupling among different species.

As a rule of thumb, if DM produces mostly electromagnetically interacting particles (high energy charged leptons and photons), the dominant effect is an increased photo-dissociation of light nuclei. Consequently, the abundance of ${}^4\text{He}$ decreases. While deuterium is initially destroyed as well, it is eventually overproduced, due to the complicated interplay in the network of reactions. Similarly, ${}^3\text{He}$ is also overproduced.

If DM primarily produces high-energy hadrons (e.g., through hadronic annihilation or decay channels), several effects can be at play. For instance, high-energy $\pi^{\pm}$ may convert protons into neutrons via the energetically favored reaction $\pi^- + p \to \pi^0 + n$. This increases the $n/p$ ratio, subsequently raising the final ${}^4\text{He}$ yield. Similarly, anti-nucleons preferentially annihilate with protons and as a result increase the $n/p$ ratio; energetic neutrons and protons instead destroy ${}^4\text{He}$ via spallation,... Generically, DM models will predict nonzero annihilation probabilities into both electromagnetic and hadronic final states, so that the effect on BBN will be a combination of the above processes.

If DM produces neutrinos, they can in principle scatter on the plasma, producing charged leptons, and therefore contributing to the electromagnetic component. However, the efficiency of this process is negligible in practice.

A careful comparison between the modified yields and the primordial element measurements allows to place the constraints on $s$-wave annihilating DM reported in fig. 6.14. These constraints are, however, largely less stringent than other ID bounds. For instance, for annihilations into $W^+W^-$, BBN imposes a bound on annihilation cross section of $\langle \sigma v \rangle \lesssim 4\,10^{-25}\,\text{cm}^3/\text{s}$. For annihilations into leptons or quarks, the bounds are even less stringent. The quoted bounds are from Hisano et al. (2009) [479], where in each case we select the nuclear element that leads to the most stringent constraints.[44] Previous, more stringent constraints, that we do not report, are given in Jedamzik and Pospelov (2009) [479].

Next, let us consider the case of **light DM**, i.e., with a mass comparable to the temperature during BBN ($M \sim T_{\text{BBN}} \sim \text{MeV}$) or lighter, as well as the scenarios where DM is accompanied by a *dark sector* of light particles. In such scenarios, the dark sector can contribute significantly to the energy budget of the Universe and modify the predictions of BBN. BBN is sensitive to the relativistic energy density: an increased expansion rate leads to an increased temperature $T_f$ at which the $n \leftrightarrow p$ reactions freeze-out. Since the neutron-to-proton ratio scales as $n_n/n_p = e^{-(m_n - m_p)/T_f}$, higher $T_f$ leads to a larger $n_n/n_p$ and thus, in particular, to a higher ${}^4\text{He}$ yield.

If the dark sector is fully decoupled from the SM sector, and as long as the dark sector particles contributing to the extra energy density remain relativistic during BBN, then the impact on BBN can be parametrized through the deviation in the effective number of neutrino species, $\Delta N_{\text{eff}}$. This is defined as

---

[44]Note that for $\mu^+\mu^-$ channel we use the $e^+e^-$ bound from Hisano, Moroi et al. (2009), which is justified since the important parameters for electromagnetically interacting particles is the total energy injected rather than the particle species.

the number of neutrino species that would lead to the same energy density as the dark sector particles under consideration (see eq. (C.27) in App. C.3). DM contributes as $\Delta N_{\text{eff}} \sim (T_{\text{DM}}/T)^4$ where $T_{\text{DM}}$ is the temperature of the dark sector. The Standard Model predicts $N_{\text{eff}} = 3.0440$ [481], where the difference from 3 species arises due to non-instantaneous neutrino decoupling and other subtle effects.

If the dark sector particles have MeV-scale mass, so that they are neither relativistic nor non-relativistic during BBN, and/or if the dark sector particles are coupled to the SM bath, e.g., interacting with electrons, neutrinos or both, then the effect is not easily condensed in a single parameter. A full analysis of the modified BBN equations then needs to be performed, taking into account in particular the modifications to the neutrino decoupling temperature (which, in the Standard Model, happens at $T_{\nu\text{fo}} \approx 2.5$ MeV). Annihilating DM particles undergo thermal freeze-out at $T \sim M/20$, so DM particles with mass $M \lesssim (1-20)\,T_{\nu\text{fo}}$ could give large effects as long as $\Gamma \sim T_{\nu\text{fo}}^3 \langle \sigma v \rangle \gtrsim H \sim T_{\nu\text{fo}}^2/M_{\text{Pl}}$ allowing to probe DM that annihilates in $e\bar{e}$ or $\nu\bar{\nu}$ with cross section $\langle \sigma v \rangle \gtrsim 1/T_{\nu\text{fo}}M_{\text{Pl}} \sim 10^{-42}\,\text{cm}^2$, of typical weak interaction size at $E \sim T_{\nu\text{fo}}$. DM particles that decay before $t_{\nu\text{fo}} \sim$ s are also tested. The detailed analysis of Sabti et al. (2020) [479] find that, if MeV-scale DM couples to electrons and photons, BBN sets a lower bound on the DM mass of the order $M \gtrsim 0.4$ MeV, under broad assumptions. If DM couples to neutrinos too, the lower bound is $M \gtrsim 1.2$ MeV. A similar previous work by Depta et al. (2019) finds $M \gtrsim 7$ MeV for both cases, under slightly more stringent assumptions. The bounds become somewhat stronger ($M \gtrsim 10$ MeV) if Planck/CMB data are included in the analysis. In fig. 5.16 and fig. 6.16 we report these constraints as $M > T_{\nu\text{fo}}$ to fix the ideas.

BBN bounds on DM annihilation rates become much weaker if DM particles are heavier than $M \gtrsim$ few $T_f \sim 20$ MeV, as already discussed above for weak-scale DM. Focusing on an example where the energy is injected via electromagnetically-interacting annihilation products, the bounds are shown in fig. 6.16. For $M \approx 200$ MeV, the constraint is $\langle \sigma v \rangle \lesssim 2\,10^{-25}\,\text{cm}^3/\text{s}$, as reported by Depta et al. (2019) [479].

Finally, let us mention that **DM can also act as a catalyst** during BBN, i.e., it can enhance the yields of light elements without being directly involved in the reaction. Specifically, between 2006 and 2008, a series of papers explored the different scenarios in detail [482]. For example, if DM is accompanied by a negatively charged state that is present in the bath during BBN (as is the case for DM that is part of an electroweak multiplet, like in supersymmetric DM), such a charged state can bind with positively charged nuclei. This reduces the Coulomb barrier between nuclei, leading to a faster fusion and a higher yield of the final product, e.g., $^6$Li.[45] Other, more subtle effects, and different scenarios, have also been explored.[46]

## 6.13   Indirect detection: status

In this chapter, we so far introduced the foundational concepts in indirect searches for DM. We are now in a position to discuss the current status. We will first review the relevant experiments and techniques. The outcome of the vast experimental effort are the measurements of cosmic ray spectra across many different energy scales. By assuming that these are compatible with the astrophysical backgrounds, the constraints on DM are derived, mostly from gamma-rays and neutrinos, but also from charged cosmic rays. Apparent anomalies that have been tentatively interpreted as indirect detection of DM are discussed in section 8.2.

---

[45] In this context, DM has been proposed as a solution to one of the *Lithium problems*, i.e., for the fact that the measured abundance of $^6$Li is higher than the predictions of standard BBN [483].

[46] Another Lithium problem has to do with the fact that the measured abundance of $^7$Li is lower than the prediction of standard BBN. Since $^7$Li does not seem to be destroyed in stars — its abundance remains fairly consistent across stars with diverse metallicity and evolutionary histories (commonly referred to as the Spite plateau) — a mechanism is needed to suppress the production of $^7$Li in BBN. A variation of the catalysis mechanism can achieve this.

## 6.13.1   Indirect detection: experiments

Table 6.2 lists many experiments relevant for DM identification. This includes past experiments (dating back to the 1970s, if still relevant), current experiments, as well as future or planned ones. For upcoming experiments, we provide an overview of anticipated timescales and capabilities, though it is worth noting that these may change. The main types of detectors for different particle species are as follows:

○ *High-energy photons (γ-rays)* are observed using three main techniques.

–  Experiments in space (such as FERMI or AMS-02) detect γ-rays directly via pair conversions in the instrument. They scan the sky continuously and have typically a large field of view.

–  Imaging atmospheric Čerenkov telescopes (IACT, such as HESS, MAGIC, VERITAS and the future CTA) detect the Čerenkov light produced in the air by the shower of particles when an energetic γ-ray collides with the top of the atmosphere. They operate only on moonless clear nights and have to point in a narrow field of view. However, these ground-based experiments are sensitive to higher energies (typically around the TeV) than space experiments (limited by the size of the instruments).

–  Air Čerenkov telescopes consist of large detectors situated on the ground, where they collect particles from atmospheric cascades. HAWC and MILAGRO, for instance, consist of large tanks filled with water. They detect passing particles via the emitted Čerenkov radiation. They have almost continuous full sky view and are sensitive to very high energies, indicatively from TeV to PeV.

○ *Low-energy photons (X-rays, radio waves)* are detected by dedicated instruments. *X*-ray devices (such as CHANDRA, INTEGRAL or EROSITA) need to be deployed on balloons, rockets, or in space. For the purposes of DM searches they typically offer good spatial accuracy and exquisite energy resolution. Radio telescopes are ground-based instruments and typically offer even better spatial resolution.

○ *Charged cosmic rays (positrons, electrons, anti-protons)* are detected in space in a manner analogous to γ-rays. Using tracking and calorimetry, the detectors determine the identity of the particle and measure its energy. Tracking and calorimetry remain effective up to a certain maximal energy, largely dictated by the size of the apparatus, which, on the other hand, is limited by the fact that the detector must be sent into space. Coupled with the fact that the fluxes at higher energies are suppressed, this explains why the spectra are measured only up to a certain maximal energy, as shown in fig. 6.12. If the experiment is equipped with a magnet (like PAMELA or AMS), it can distinguish between particle charges, allowing for differentiation between particles and antiparticles based on the unique curvature of their respective tracks. This capability is crucial for DM searches because, generally, the astrophysical background of anti-particles is smaller than the one of particles. Experiments lacking a magnet (like CALET or DAMPE) are limited to measuring the total flux, e.g., $e^+ + e^-$. FERMI, however, has been able to successfully use the magnetic field of the Earth for this purpose. Above about 1 TeV, the spectra of electrons and positrons have also been measured on Earth by the IACT detectors (such as HESS and VERITAS).

○ *Anti-deuterons (and heavier anti-nuclei)* are detected in space using methods similar to those for other charged particles: they can be differentiated from the much more abundant anti-protons, e.g., in AMS-02, thanks to different charge-to-mass ratio. The dedicated balloon experiment GAPS [530] aims to detect anti-deuterons employing a novel technique: the approach involves slowing down the $\bar{d}$ nucleus, capturing it within the detector to form an exotic atom, which then annihilates, emitting characteristic *X*-ray and pion radiation. Like previous balloon missions, GAPS will be

| Experiment | Location | Operation | Technology | Main focus | Energy range | Home | Ref. |
|---|---|---|---|---|---|---|---|
| Heao-1 | satellite | 1977 → 1979 | X-ray detectors | $X/\gamma$-rays | 0.2 keV − 10 MeV | web | [484] |
| Baksan | Russia | 1978 → | scintillation | neutrinos | 1 GeV − 1 TeV | web | [485] |
| Rosat | satellite | 1990 → 1999 | X-ray detectors | $X$-rays | 0.1 − 2.5 keV | web | [486] |
| Comptel | satellite | 1991 → 2000 | HEP detectors | $\gamma$-rays | 1 − 30 MeV | web | [487] |
| Egret | satellite | 1991 → 2000 | HEP detectors | $\gamma$-rays | 30 MeV − 30 GeV | web | [488] |
| Cangaroo | Australia | 1992 → 2012 | air Čerenkov | $\gamma$-rays | 200 GeV − 3 TeV | web | [489] |
| Heat | balloon | 1994, 1995 | HEP detectors | $e^-$ & $e^+$ | 1 − 100 GeV | − | [490] |
| Super-Kam. | Japan | 1996 → | water Čerenkov | neutrinos | few MeV − $\gtrsim$100 GeV | web | [491] |
| Amanda | South Pole | 1996 → 2005 | ice Čerenkov | neutrinos | 50 GeV − $\gtrsim$10 TeV | web | [492] |
| Ams-01 | Space shuttle | 1998 | HEP detectors | charged CRs | 0.1 − 200 GeV | web | [493] |
| Baikal-NT | Siberia | 1998 → | water Čerenkov | neutrinos | 10 GeV − few TeV | web | [494] |
| Chandra | satellite | 1999 → | X-ray detectors | $X$-rays | 0.1 − 100 keV | web | [495] |
| Xmm-Newton | satellite | 2000 → | X-ray detectors | $X$-rays | 0.15 − 15 keV | web | [496] |
| Milagro | New Mexico | 2001 → 2008 | water Čerenkov | $\gamma$-rays | 100 GeV − 100 TeV | web | [497] |
| Integral | satellite | 2002 → 2025 | HEP detectors | X-/$\gamma$-rays | 15 keV − 20 MeV | web | [498] |
| Hess | Namibia | 2003 → | air Čerenkov | $\gamma$-rays | 30 GeV − 100 TeV | web | [499] |
| Veritas | Arizona | 2004 → | air Čerenkov | $\gamma$-rays | 50 GeV − 50 TeV | web | [500] |
| Magic | Canary Islands | 2004 → | air Čerenkov | $\gamma$-rays | 30 GeV − 100 TeV | web | [501] |
| Swift | satellite | 2004 → | X-ray detectors | $X$-rays | 0.2 − 10 keV | web | [502] |
| Cream | Antarctic balloon | 2004 → 2010 | HEP detectors | CR nuclei | 10 GeV − 100 TeV | web | [503] |
| Suzaku | satellite | 2005 → 2015 | X-ray detectors | $X$-rays | 0.2 − 600 keV | web | [504] |
| Icecube | South Pole | (2005) 2010 → | ice Čerenkov | neutrinos | $\gtrsim$ 100 GeV | web | [505] |
| Pao (Auger) | Argentina | 2005 → | air shower/water Č. | UHECR | >100 PeV | web | [506] |
| Anita | Antarctic balloon | 2006 → | Askaryan effect | neutrinos | 0.1 − 100 EeV | web | [507] |
| Pamela | satellite | 2006 → 2016 | HEP detectors | charged CRs | 50 MeV − 1 TeV | web | [508] |
| Fermi | satellite | 2008 → | HEP detectors | $\gamma$-rays | 20 MeV − 500 GeV | web | [509] |
| Antares | French riviera | 2008 → 2021 | water Čerenkov | neutrinos | 10 GeV − 1 PeV | web | [510] |
| Ams-02 | ISS | 2011 → | HEP detectors | charged CRs | 500 MeV − 2 TeV | web | [511] |
| NuStar | satellite | 2012 → | X-ray detectors | $X$-rays | 3 − 79 keV | web | [512] |
| Taiga | Siberia | ∼2012 → | air Čerenkov | $\gamma$-rays/CRs | few TeV − 100 PeV | web | [513] |
| Hawc | Mexico | 2014 → | water Čerenkov | $\gamma$-rays | 100 GeV − 100 TeV | web | [514] |
| Tibet AS | Tibet | 2014 → | air shower/water Č. | $\gamma$-rays/CRs | ≥ 100 TeV | web | [515] |
| Calet | ISS | 2015 → | HEP detectors | charged CRs | 1 GeV − 20 TeV | web | [516] |
| Hitomi | satellite | 2016 | X-ray detectors | $X$-rays | 0.3 − 80 keV | web | [517] |
| Dampe | satellite | 2016 → | HEP detectors | charged CRs | 5 GeV − 10 TeV | web | [518] |
| Cosi-Spb | balloon | 2016 | Compton telescope | $\gamma$-rays | 0.2 − 5 MeV | web | [519] |
| Hxmt | satellite | 2017 → | X-ray detectors | $X/\gamma$-rays | 1 − 250 keV | web | [520] |
| Iss-Cream | ISS | 2017 → | HEP detectors | charged CRs | 10 GeV − 100 TeV | web | [521] |
| Mace | Himalaya | 2017 → | air Čerenkov | $\gamma$-rays | 40 GeV − 20 TeV | − | [522] |
| Micro-X | New Mexico | 2018 | X-ray detectors | $X$-rays | 0.2 − 3 keV | web | [523] |
| eRosita | satellite | 2019 → | X-ray detectors | $X/\gamma$-rays | 0.3 − 10 KeV | web | [524] |
| Lhaaso | China | 2020 → | air shower/water Č. | $\gamma$-rays/CRs | 100 GeV − EeV | web | [525] |
| Km3Net | Mediterranean | 2022 → | water Čerenkov | neutrinos | $\gtrsim$ 1 TeV | web | [526] |
| Cta | North+South | 2020s?+? | air Čerenkov | $\gamma$-rays | 50 GeV − 50 TeV | web | [527] |
| Xrism | satellite | 2023 → | X-ray detectors | $X$-rays | 0.3 − 13 keV | web | [528] |
| Baikal-Gvd | Siberia | 2023 → | water Čerenkov | neutrinos | 100 GeV − few PeV | web | [529] |
| Gaps | Antarctic balloon | 2026? | nuclear physics | $\bar{d}$ | 0.1 − 0.3 GeV/n | web | [530] |
| Adept | balloon | 2026? | HEP detectors | $\gamma$-rays | 5 − 200 MeV | − | [531] |
| Cosi | satellite | 2026? | Compton telescope | $\gamma$-rays | 0.2 − 5 MeV | web | [532] |
| Hyper-Kam. | Japan | 2028? | water Čerenkov | neutrinos | few MeV − $\gtrsim$100 GeV | web | [533] |
| Gamma-400 | satellite | 2020s? | HEP detectors | $\gamma$-rays | 100 MeV − 3 TeV | web | [534] |
| Herd | Chinese SS | 2020s? | HEP detectors | charged CRs | 50 GeV − 1 PeV | web | [535] |
| Ska | S.Africa+Australia | 2020s? | radio telescope | radio | 50 MHz − 30 GHz | web | [536] |
| Ino-Ical | India | 2020s? | calorimeter | neutrinos | 1 − 100 GeV | web | [537] |
| Amego | satellite | late 2020s? | HEP detectors | $\gamma$-rays | 0.2 MeV − 10 GeV | web | [538] |
| Apt | satellite | late 2020s? | HEP detectors | $\gamma$-rays | 60 MeV − 1 TeV | − | [539] |
| Dune | USA | 2030? | liquid Argon | neutrinos | $\gtrsim$ 10 MeV | web | [540] |
| Athena | satellite | early 2030s? | X-ray detectors | $X/\gamma$-rays | 0.2 − 12 keV | web | [541] |
| as-/e-Astrogam | satellite | 2030s? | HEP detectors | $\gamma$-rays | 0.1 MeV − 3 GeV | − | [542] |
| Grand | high altitude deserts | 2030s? | radio telescopes | neutrinos | 100 PeV − 100 EeV | web | [543] |
| Aladino | L2 point? | 2035? | HEP detectors | charged CRs | → 10 TeV | − | [544] |
| Ams-100 | L2 point | 2039? | HEP detectors | charged CRs | sub-GeV − 10 TeV | − | [545] |
| Gecco | satellite | proposed | HEP detectors | $X/\gamma$-rays | 100 keV − 10 MeV | − | [546] |
| Grams | balloon/satellite | proposed | LAr detector | $\gamma$-rays/$\bar{d}$ | 200 keV − 200 MeV | − | [547] |
| Mast | satellite | proposed | LAr satellite | $\gamma$-rays | 100 MeV − 1 TeV | − | [548] |
| Swgo | South America | proposed | water Čerenkov | $\gamma$-rays | 100 GeV − 1 PeV | web | [549] |
| P-One | Pacific NW | proposed | water Čerenkov | neutrinos | 10 TeV − 10 PeV | web | [550] |
| Trident | South China sea | proposed | water Čerenkov | neutrinos | 1 TeV − 10 PeV | web | [551] |

Table 6.2: **Main experiments** *with some relevance for* **Indirect Detection** *as of early 2025. The experiments are ordered by the year of commissioning. Additional links to the homepages of these experiments can be accessed here. The references in the last column are mostly to the documents describing the experiments (technical design reports, reviews, science cases, proposals,...) and not to the scientific results. We gratefully acknowledge the contribution of Aloïse Dijoux in compiling this table.*

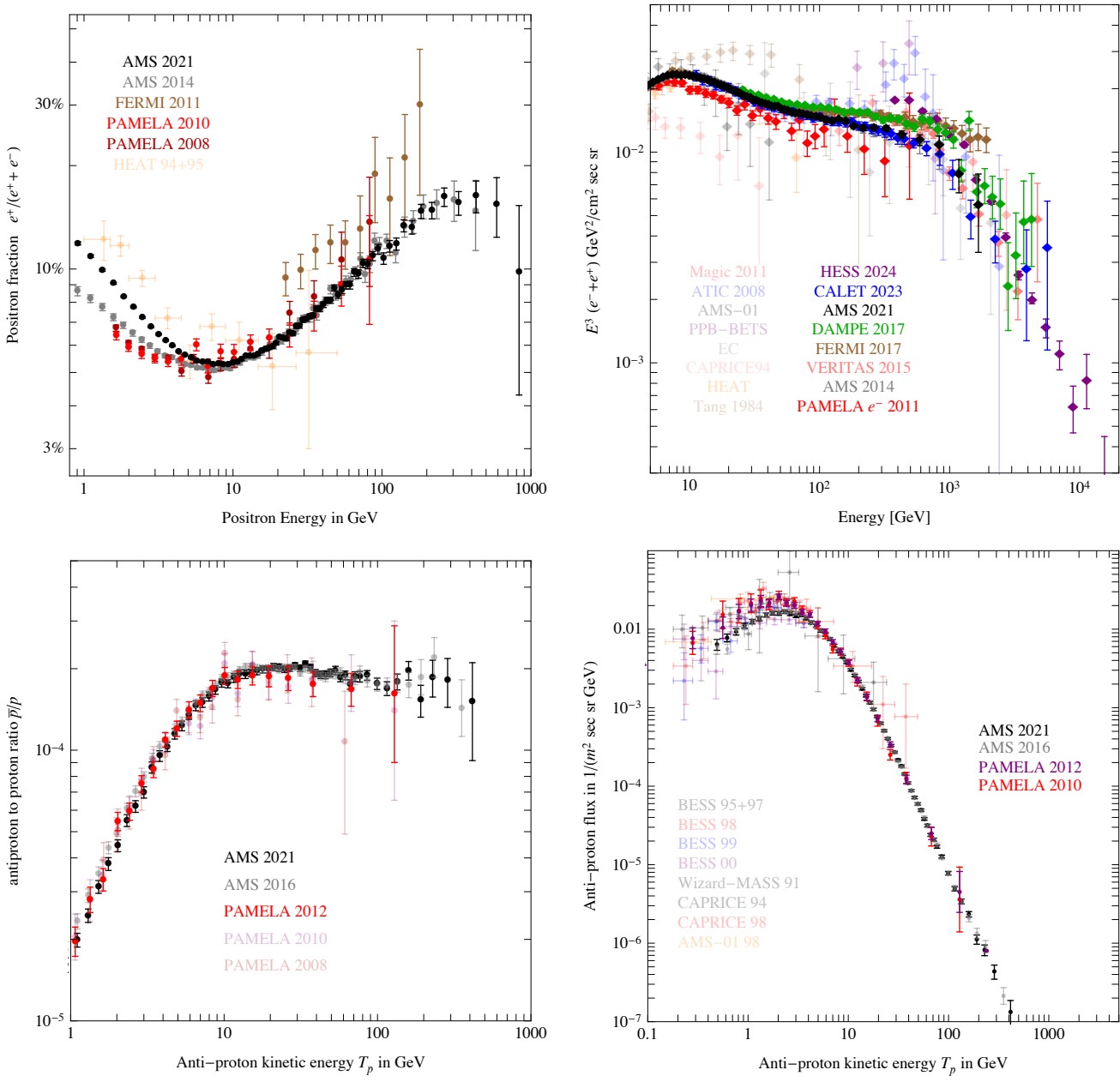

Figure 6.12: *A compilation of both historical and recent measurements of **charged cosmic rays**, relevant for weak-scale DM indirect detection. Top left: positron fraction. Top right: electron and positron ('all lepton') flux. Bottom left: anti-proton to proton ratio. Bottom right: anti-proton flux. References are given in table 6.4.*

flown over Antarctica to maximize the signal — low energy charged particles are deflected by the terrestrial magnetic field at lower latitudes. For a comprehensive overview see [552].

- ○ *Neutrinos* are detected using large Čerenkov detectors situated underground (or under-ice or under-water). The detection is via showers of secondary particles generated when neutrinos interact with the material inside the instrumented volume or its surroundings. Among these secondary particles, charged ones, in particular muons, emit Čerenkov light when traversing the detector and thus their energies and directions can be measured (these are partially correlated with those of the parent neutrino). The main background for this search is due to the large flux of cosmic muons coming from the atmosphere above the detector. To mitigate this the experiments typically have to select only up-going tracks, i.e., due to neutrinos that have crossed the entire Earth and interacted either within or just below the instrumented volume.

### 6.13.2  Indirect detection: current constraints

A large number of searches for DM annihilation or decay signals turned up empty handed. Therefore, these measurements merely set upper bounds on DM annihilation cross section or decay rate. These limits are obtained by requiring that the flux from DM, possibly combined with the known or presumed astrophysical background, does not exceed the measured flux. Results are conveniently presented in the planes $(M, \langle\sigma v\rangle_{\mathrm{ann}})$ and $(M, \tau)$, where for annihilation (decay) the regions above (below) the limiting lines are ruled out.

#### Constraints on DM with weak-scale mass

We first consider weak-scale DM, with a mass roughly in the range GeV to 100 TeV, which has been the focus of extensive searches in the past decades. Fig. 6.14 and 6.15 provide a selection of the most relevant bounds, organized by DM annihilation channel and by SM messenger particle species. The CMB constraints are discussed in section 6.11.2 and the BBN ones in section 6.12.

For DM annihilations, the relevant benchmark is the thermal cross-section derived in eq. (4.13) and plotted in detail in fig. 4.1 (left). As discussed in section 4.1, this cross-section corresponds to the annihilation rate that, within a large class of minimal models, reproduces the observed DM relic abundance through a thermal freeze-out process. In the case of decays, there is no analogous benchmark decay rate.

The annihilation constraints scale with the DM mass roughly as $\langle\sigma v\rangle/M \approx$ constant (except the constraints on neutrino fluxes, see below). This scaling can be understood qualitatively: since the DM number abundance scales as $1/M$, the flux of annihilation products (charged cosmic rays or gamma rays) scales as $M^{-2}$, see eq. (6.4). The total amount of injected products, on the other hand, is proportional to $M$, since all the energy of the annihilated DM particles transforms into cosmic rays, hence the bound on $\langle\sigma v\rangle$ scales as $\propto M$. This scaling is well respected by the CMB constraints (see the discussion in section 6.11), for which the total energy injected in the DM annihilation is the only relevant parameter. For charged cosmic rays and gamma rays, the shape of the spectrum and the propagation effects also play a role, making the dependence on $M$ less pronounced. For DM decay rates, the constraints remain constant with respect to $M$, because the flux of annihilation products scales as $M^{-1}$.

**Gamma-rays**. Gamma-ray searches for DM signals have been pursued since the very beginning of indirect detection, in a variety of targets and energy ranges. In table 6.3 we list the main (negative) results, both for DM annihilations and decays. For brevity, we focus mostly on the 'official' results from the experimental collaborations rather than on the works by independent groups of theorists. A selection of the bounds is reproduced in fig. 6.13.[47] Note that some of the constraints have been rescaled from their

---

[47]For a recent exhaustive list of searches see Hütten and Kerszberg (2022) in [1].

|  | GC / GCH | MW halo | Dwarfs | Clusters | Cosmological | other |
|---|---|---|---|---|---|---|
| **DM annihilation: γ-ray searches** | | | | | | |
| continuum | HESS [553–556]<br>(FERMI [566])<br>Abazajian et al. [572] | FERMI [557]<br>HAWC [567, 568] | MAGIC [558–560]<br>HESS [569, 570]<br>FERMI [573]<br>FERMI+MAGIC [576]<br>FERMI-DES [577]<br>HAWC [579]<br>VERITAS [580]<br>FERMI+IACTS+<br>+HAWC [581]<br>McDaniel et al. [582] | FERMI [561–563]<br>HESS [571]<br>VERITAS [574] | FERMI [564] | FERMI [565]<br>(dark satellites)<br>HESS [575]<br>(GloCs)<br>VERITAS [578]<br>(subhalos) |
| lines | HESS [583, 584] | FERMI [585] | MAGIC [558]<br>HESS [570, 587]<br>VERITAS [580]<br>HAWC [588] | FERMI [563, 586] | | |
| **DM annihilation: neutrino searches** | | | | | | |
| continuum | ANTARES [589–591]<br>ICECUBE [596, 597]<br>SUPERK [600, 601]<br>BAIKAL [602]<br>ANTARES+<br>ICECUBE [603]<br>ANTARES (heavy/<br>secluded DM) [604] | ICECUBE [592–594]<br>ANTARES [598] | ICECUBE [595]<br>BAIKAL [599] | ICECUBE [595] | | |
| lines | ANTARES [591, 598]<br>ICECUBE [596, 597]<br>SUPERK [600, 601]<br>BAIKAL [602] | ICECUBE [592–594] | ICECUBE [595]<br>BAIKAL [599] | | | |
| **DM decay: γ-ray searches** | | | | | | |
| continuum | Cohen et al. [605]<br>Esmaili et al. [609] | FERMI [557]<br>HAWC [567, 568] | VERITAS [606]<br>MAGIC [558]<br>HAWC [611] | FERMI [607]<br>MAGIC [610]<br>HAWC [612] | Cirelli et al. [608] | |
| lines | | FERMI [585] | MAGIC [558] | | | |

**DM decay: neutrino searches:** ICECUBE [594, 613–615]

Table 6.3:  *A collection of the currently most relevant **γ-ray and neutrino searches for weak-scale DM**, as produced by different (mostly experimental) collaborations. The top tables pertain to the annihilating DM, the bottom one to decaying DM. The targets correspond to the broad classification presented in section 6.2.*

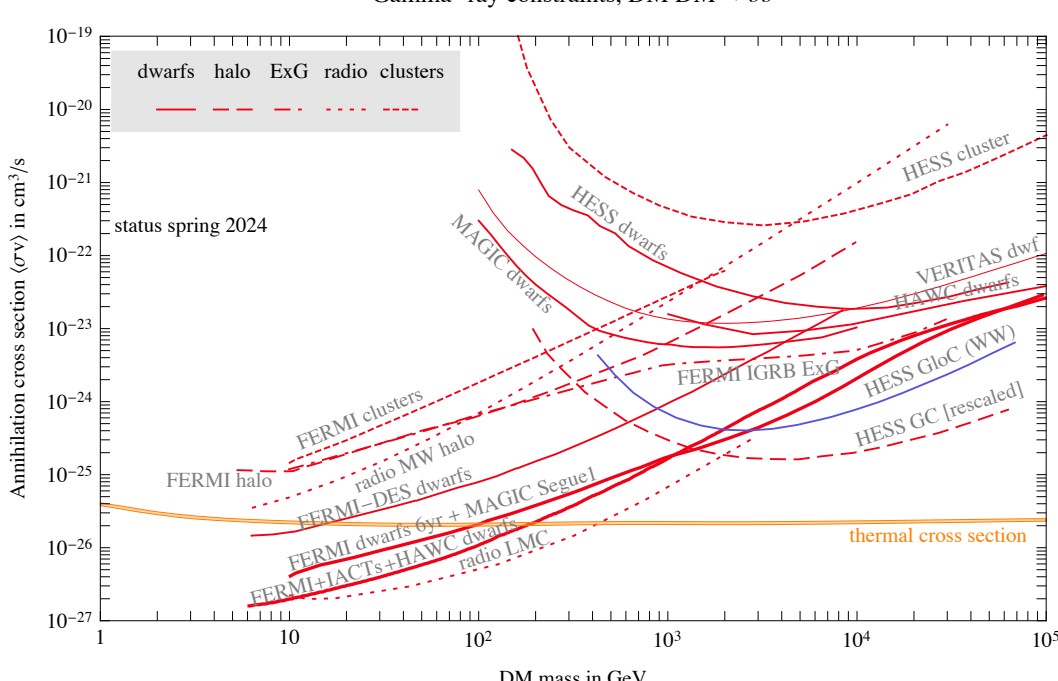

Figure 6.13: ***Bounds on weak-scale DM annihilations*** *imposed by different **gamma-ray** (and radio) observations assuming a typical hadronic DM annihilation channel, into $b\bar{b}$. Certain constraints have been rescaled with respect to those published in the literature, in order to correct for the different choices of DM profiles: when possible, they have been brought to the standard NFW profile. The almost horizontal orange thin band is the thermal cross section of eq. (4.13), also plotted in fig. 4.1 (left). We gratefully acknowledge the contribution of Aloïse Dijoux in compiling these results.*

originally published values, in order to correct for the different choices of DM profiles: when possible, they have been brought to the standard NFW profile.

For DM masses up to about 1 TeV the most stringent constraints come from dwarf galaxies. For heavier DM, the constraints derived from IACT observations of the GC, notably HESS, take over.

**Radio-waves**. Radio observations have long been employed to set constraints on the synchrotron emissions from DM [454]. Historically, observations focused on small regions near the GC, since the expected large DM density and the presence of a strong magnetic field guarantee significant emissions in this region. However, uncertainties are also largest around the GC, both regarding the DM distribution, the astrophysical environments and the background emissions. Later on, several studies shifted the focus to much larger regions of the galactic halo (or even the whole galactic halo), which can provide more robust results. Outer galaxies and galaxy clusters have also been considered, as well as isotropic signals of extragalactic origin. In fig.s 6.13, 6.14 and 6.15, we report a few selected bounds, selecting the most conservative estimates. Notably, the recent constraints derived in Regis et al. (2021) [454] from observations of the Large Magellanic Cloud, if confirmed, are very stringent.

Manconi et al. (2022) [454] considered the *polarization* of the radio-waves emitted by $e^+e^-$ from DM annihilations, while these gyrated in the galactic magnetic field, and compared it with the polarization maps produced by PLANCK. This results in stringent constraints on leptonic DM annihilation channels, however, assumptions regarding the galactic magnetic field can change the results by approximately an order of magnitude.

**Anti-protons (and electrons or positrons)**. Measurements of charged cosmic rays, a compilation

Figure 6.14: *Summary chart of the **current most stringent bounds on weak-scale DM annihilations** into $\mu^+\mu^-$ (green), $b\bar{b}$ (red) or $W^+W^-$ (blue), from different searches, as discussed in the main text. Some constraints are rescaled with respect to the literature, to correct for the different choices of DM profiles. The almost horizontal orange thin band is the thermal cross section of eq. (4.13), also plotted in fig. 4.1 (left).*

of which is given in figure 6.12 and table 6.4, can also be used to impose competitive constraints. The anti-proton bounds reported in fig. 6.14 and 6.15 were derived in [664]. They are based on the 2016 results of AMS-02, but do not change significantly if the 2021 data are used instead. The reported bounds are significantly more stringent than previous constraints [665] for DM masses larger than 200 GeV and less stringent for lower DM masses. Notably, some authors have recently identified a possible excess at low DM masses in the same data, as discussed in section 8.2.9.

Some authors have also derived bounds from electrons or positrons (not reported in the figure) [666]. Given that these channels exhibit clear excesses (see section 8.2.8), the strategy is to attribute these excesses to astrophysical sources, and then search for any additional features in the spectrua that would be produced by DM. This strategy is possible for DM annihilating or decaying into a leptonic channel, since this produces a distinctive sharp feature.

**Anti-deuterons**. The only current upper bound on the flux of cosmic anti-deuterons comes from the BESS experiment [658, 659] and is too weak to give any significant constraints on annihilating or decaying DM. The planned sensitivities of AMS and GAPS, though, may allow to test interesting regions of the DM parameter space, for 100 GeV DM even down to the thermal cross section, in optimal conditions [429, 552, 667]. An advantage of anti-deuteron searches is that the astrophysical background and DM signal peak at different energy ranges. Specifically, while the astrophysical background typically peaks at a few GeV per nucleon, the DM signal is expected to peak at energies below 1 GeV/n, so that the search for a DM $\bar{d}$ signal can be considered as essentially background-free. The underlying reason is that the astrophysical $\bar{d}$ are produced in spallation of high-energy cosmic rays on the interstellar gas: the kinematics is such that low energy $\bar{d}$ are produced only very rarely. The DM $\bar{d}$'s are, in contrast,

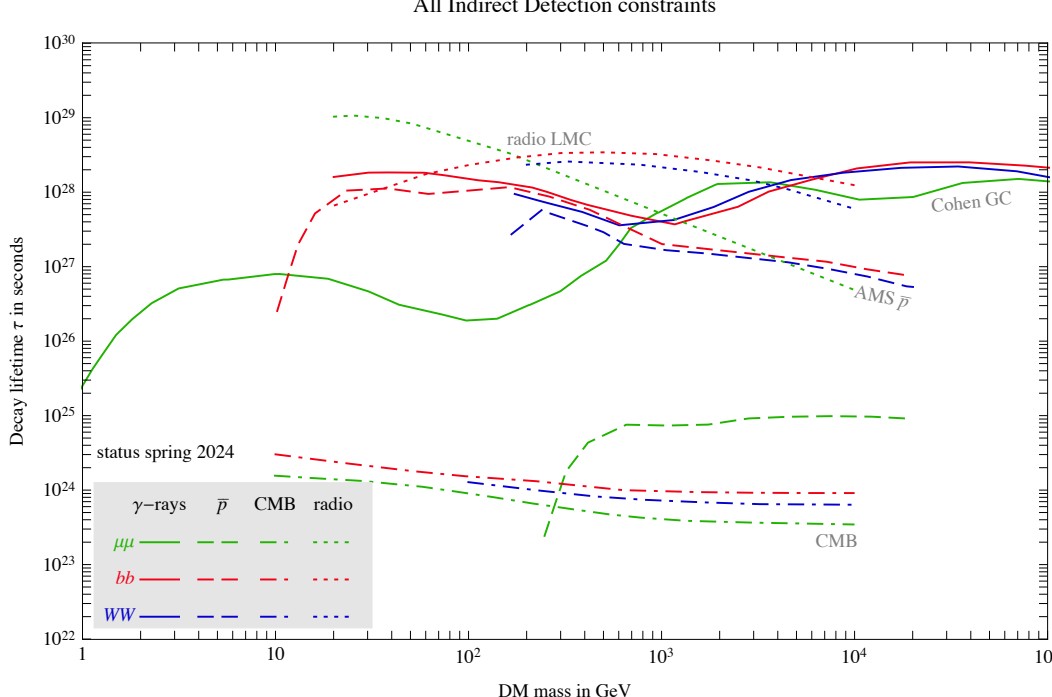

Figure 6.15: *Summary plot of the **most stringent current bounds on decaying weak-scale DM**, in different channels and from different searches, as discussed in the main text.*

| CR species | Experiments |
|---|---|
| Positron fraction $e^+/(e^+ + e^-)$ | HEAT 1997 [616] − PAMELA 2008 [617] − PAMELA 2010 [618] − FERMI 2011 [619] − AMS 2013 [620] − PAMELA 2013 [621] − AMS 2014 [622] − AMS 2019 [623] |
| All leptons $(e^+ + e^-)$ | HESS 2009 [624] − FERMI 2010 [625] − MAGIC 2011 [626] − AMS 2014 [627] − VOYAGER1 2015 [628] − VERITAS 2015 [629] − FERMI 2017 [630] − HESS 2017 [631] − DAMPE 2017 [632] − CALET 2017 [633] − CALET 2018 [634] − VERITAS 2018 [635] − AMS 2019 [623] − AMS 2021 [636] − CALET 2023 [637] − HESS 2024 [638] |
| Positrons $e^+$ | FERMI 2011 [619] − PAMELA 2013 [621] − AMS 2014 [639] − AMS 2018 [640] − AMS 2019 [641] |
| Electrons $e^-$ | PAMELA 2011 [642] − FERMI 2011 [619] − AMS 2014 [639] − AMS 2018 [640] − AMS 2019 [623] |
| anti-proton/proton ratio $\bar{p}/p$ | PAMELA 2008/10/12 [643] − ARGO-YBJ 2012 [644] − AMS 2016 [645] |
| anti-protons $\bar{p}$ | WIZARD-MASS 1991 [646] − CAPRICE 1994 [647] − BESS 1996-97 [648] − BESS 1998 [649] − CAPRICE 1999 [650] − BESS 1999-2000 [651] − AMS01 2001 [652] − PAMELA 2010/12 [643] − BESS 2012 [653] − AMS 2016 [645] − AMS 2021 [636] |
| Protons $p$ | PAMELA 2011 [654] − VOYAGER1 2015 [628] − AMS 2015 [655] − CALET 2019 [656] − DAMPE 2019 [657] − AMS 2021 [636] |
| Antideuterons $\bar{d}$ | BESS 2005 [658] − BESS-POLAR II 2023 [659] |
| Antihelium $\overline{He}$ | AMS-01 1999 [660] − PAMELA 2011 [661] − BESS-POLAR II 2012 [662] |

Table 6.4: ***Recent data on charged cosmic rays*** *with relevance for weak-scale DM indirect detection. An exhaustive list of data can be found on the Cosmic-Ray database [663].*

produced from a coalescence of $\bar{p}$ and $\bar{n}$ from essentially-at-rest DM particles and are not subject to the same kinematical constraints. For a recent detailed discussions of the light anti-nuclei astrophysical backgrounds see [668].

**Anti-helium**. Upper bounds on the flux of cosmic anti-helium nuclei come from AMS-01 [660], PAMELA [661] and BESS-POLAR II [662]. They are much higher that than the flux predicted by DM and by astrophysics. See however the detailed discussion in section 8.2.10 concerning a possible anomaly in this channel.

**Neutrinos**. All current experiments (ICECUBE, ANTARES, SUPERKAMIOKANDE and BAIKAL) impose constraints from the non-observation of high-energy neutrino fluxes from the Galactic Halo and Galactic Center. References for these experiments can be found in table 6.3, and the most significant bounds are depicted in fig.s 6.14 and 6.15.[48] Quantitatively, the constraints derived from neutrino flux are somewhat weaker than the $\gamma$-ray bounds discussed above (both in the case of DM annihilation and decay). Nevertheless, they offer an advantage: they are less dependent on the DM particle mass. This is because neutrino cross sections grow with energy, so that the higher energy neutrinos coming from annihilations of heavier DM have a higher probability for detection in the material of a neutrino telescope, which partly compensates for the reduced flux. Neutrino constraints, both for annihilation and decay, are competitive with $\gamma$-ray constraints for DM masses above several TeV. Arguelles et al. (2019) [669] compiled current bounds and future sensitivities also for very small and very large DM masses. A robust upper bound on the total annihilation cross-section, of the order of $\sigma v_{\rm rel} \lesssim 10^{-21}\,{\rm cm}^3/\sec$ for $1\,{\rm GeV} \lesssim M \lesssim 100\,{\rm TeV}$, can be imposed focusing on the least detectable messengers, namely neutrinos, and comparing the predicted flux with the atmospheric neutrino measurements [670].

**Invisible radiation**. The weakest bounds arise if DM annihilates or decays into invisible radiation, such as gravitons or hypothetical extra particles much lighter than $M/2$. The bound on the DM lifetime is [671]

$$\tau/f \gtrsim 250\,{\rm Gyr}, \tag{6.80}$$

where the factor $f \leq 1$ accounts for the possibility that only a fraction $f$ of DM decays into invisible relativistic particles (this decaying component of DM has a life-time $\tau$). The above bound implies that up to today at most $\approx 5\%$ of DM decayed or annihilated in this way.

**Other constraints**. The observation of the absorption of 21cm radiation by the EDGES experiment (see section 8.4.5) can be used to derive constraints on both annihilating and decaying DM [672].

Another strategy used to set bounds on annihilating or decaying DM are the cross-correlations among different sources or signals discussed in section 6.2.3. For annihilating DM, the cross-correlations exclude the thermal cross-section for DM with mass below $M \lesssim 20\,{\rm GeV}$, in specific configurations (see e.g. [437]). These constraints are an order of magnitude weaker than the most stringent bounds from other strategies discussed above. For decaying DM, the decay times of roughly $10^{-25-26}/$s are excluded, which is also much weaker than the bounds obtained using other probes. The projected future sensitivities of cross-correlations are, however, quite promising, possibly reaching the thermal cross-section for TeV DM, in ideal configurations.

The rough overall result is that indirect detection implies $\sigma v_{\rm rel} \lesssim 10^{-27-24}\,{\rm cm}^3/\sec$, depending on the DM mass and annihilation channel. In some cases this excludes the thermal cross section, e.g., for DM lighter than $\sim 200\,{\rm GeV}$ annihilating into $b\bar{b}$ quarks. Assuming that DM annihilates into a generic visible channel implies the lower limit $M \gtrsim 20\,{\rm GeV}$, obtained in Leane et al. (2018) [673] by marginalizing over the branching ratios. Bounds on the DM lifetime are at the $\tau \gtrsim 10^{27-28}$ seconds level, about 10 orders of magnitude longer than the age of the Universe.

---

[48]The combined ANTARES + ICECUBE limits in [603] are more stringent than the individual ones. However, they span only a limited range of DM masses, so they are not reported in the figure.

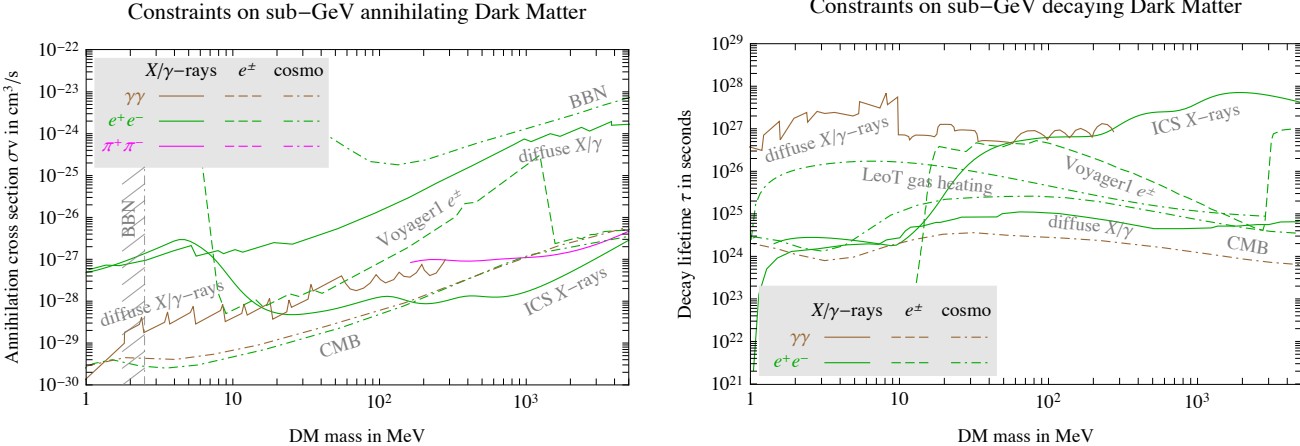

Figure 6.16:   *Summary plots of the current **bounds on sub-GeV DM** annihilations (left) or decays (right) from different searches, as discussed in the main text.*

## Constraints on sub-GeV DM

Next, let us consider DM with roughly MeV to GeV mass. The indirect detection constraints for this mass range are reported in figure 6.16 [674]. The kinematically allowed annihilation or decay channels are: $\gamma\gamma$, neutrinos, $e^+e^-$, $\mu^+\mu^-$. One can also consider DM annihilations or decays into quarks, producing $\pi^+\pi^-$ and $\pi^0\pi^0$. If DM is heavier than a few GeV, it can also produce $p\bar{p}$ pairs.

The basic principles and techniques for indirect searches in this mass range are the same as for weak-scale DM, discussed above. However, certain unique characteristics do exist:

- Regarding charged particles (primarily $e^\pm$ and potentially $\bar{p}$): particles with sub-GeV energies are not able to penetrate the heliosphere and therefore cannot be detected by detectors located in the Earth's orbit. Currently, only the VOYAGER-1 and -2 space probes have crossed the outer boundary of the heliosphere, and are thus uniquely positioned to search for these signals.

- The MeV to GeV photons (i.e., the hard $X$-rays and soft $\gamma$-rays) are in principle a powerful tool, however, there are only a few current telescopes with good sensitivities and acceptances in this energy range (see the list in table 6.2). At energies below a few MeV, INTEGRAL and CHANDRA can be used. At energies above 100 MeV FERMI-LAT takes over. In between, only the relatively old telescope COMPTEL is relevant. A number of different experiments are being proposed to fill this so-called *MeV gap*. An alternative approach is to look for secondary emissions: if sub-GeV DM annihilates or decays into channels that contain $e^\pm$, these produce inverse Compton photons in the $X$-ray energy range (see section 6.7.1), which can be tested with telescopes such as XMM-NEWTON. Currently, this actually produces some of the most stringent constraints.[49]

- DM neutrinos with MeV energies are typically overwhelmed by the solar and atmospheric neutrino backgrounds.

- The cosmological bounds remain relevant in the entire DM mass range. For annihilating DM the CMB constraints are among the most stringent ones. For decaying DM, there are relevant constraints for low DM masses which arise from requiring that there is no excessive heating in

---

[49]In the same vein, let us mention that the synchrotron secondary emission from $e^\pm$ around super-massive black holes can lead to even more stringent constraints, however under significant assumptions for the magnetic field and the DM distribution (see Chen et al. (2024) in [674]).

the gas-rich dwarf galaxy Leo T. For annihilating DM, the BBN bounds discussed in section 6.12 exclude a portion of the parameter space at small mass and another portion at large cross sections.

### Constraints on DM with mass above 100 TeV

Let us briefly discuss super-heavy DM, characterized by $M \gtrsim 100$ TeV. Such a large mass for annihilating DM requires a special mechanism to avoid the unitarity constraints discussed in section 4.1.4. For this reason most of the bounds in the literature focus on the decaying DM case. Model-independent constraints have been derived by the Hawc [567, 568, 612], Magic [610] and Lhaaso [675] collaborations, and by a number of independent groups [605, 609, 676–679] using data from other high-energy cosmic ray observatories such as Kascade, Kascade-Grande, Pao (Pierre-Auger-Observatory), Ta (Telescope Array), Casa-Mia, Tibet-AS$\gamma$ as well as Icecube. The Pao collaboration derived constraints on a specific scenario of super-heavy decaying DM coupled to ultralight sterile neutrinos [680]. The bounds on the DM lifetime reach $\tau \gtrsim 10^{29-30}$ sec, which is somewhat higher than for weak-scale DM. The Hawc [567, 568, 681] collaboration also derived constraints on annihilating super-heavy DM.[50]

## 6.14 Astrophysical searches for ultra-light DM

As discussed in section 3.4, DM could be a sub-eV boson, behaving as a classical oscillating field. On galactic and stellar scales, such light and ultra-light DM can exhibit unique phenomena, providing opportunities to probe and restrict its properties. The discussion of indirect detection constraints on light DM, focusing on axions and axion-like particles, is covered in section 10.4.5. In this section, we focus on the phenomenology primarily associated with ultra-light (fuzzy) DM, which exhibits the following features [683]:

○ Numerical simulations find that the **DM density distributions** exhibit a **core** at the centers of galaxies and dwarf galaxies (these cores are often referred to as 'solitonic', meaning that their sizes are comparable to the DM wavelength). Outside the cores, DM density transitions to the NFW profile. One can show (see N. Bar et al. (2018) in [683]) that in this case the circular velocities of the *inner* stars (within the core) are expected to be comparable to the circular velocities of the *outer* stars. This is not seen in observations, which rather find that the inner stars have smaller circular velocities. Using this observational data one can place a bound on DM mass, though a rather weak one, $M \gtrsim 10^{-21}$ eV, which is already the minimum DM mass compatible with the dwarf galaxy sizes, see eq. (3.1).

If the dwarf galaxies do have cores, their masses $M_c$ are expected to scale with the masses of the galactic halos as $M_c \propto M_{\text{halo}}^{\alpha}$, where $\alpha$ is typically found to be $1/3$ but reaches $\sim 5/9$ in some studies [683]. This predicts for dwarf satellite galaxies to have cores of kpc sizes and masses of $\sim 10^8 M_\odot$, which is in the right ballpark to potentially solve the small scale controversies in the cold DM model (see section 8.5).

○ **Interference** between different modes of the scalar field results in temporally transient **density fluctuations** roughly the size of the DM wavelength. The interference can be constructive or destructive, so that the fluctuations can exhibit as either significantly over-dense or under-dense regions.[51] For example, in regions where the field amplitude vanishes, $\psi = 0$, this defines 1-dimensional curves in space, approximately one per de Broglie volume. This phenomenon can be

---

[50]For sensitivity estimates to annihilating super-heavy DM, see also [682].

[51]While fuzzy DM leads to the suppression of small-scale structures on cosmological scales due to quantum pressure (see section 3.4), the interference can create sharp structures on very small scales.

probed via gravitational lensing. In fact, the study by Amruth et al. (2023) in [683] contends that wave DM with a particle mass of $M \sim 10^{-22}$ eV provides a superior fit to a specific strong-lensing observation when compared to a particle DM NFW profile. The phenomenon can also be probed via pulsar timing, since fluctuating gravitational potentials may oscillate at frequencies comparable to those of pulsars, allowing for sensitive probes.

These fluctuations in the gravitational potential can furthermore heat up the random motions of stars and disrupt tight conglomerations of stars such as the globular clusters. Using observations of globular clusters within the Eridanus II dSph galaxy, Marsh and Niemayer (2019) in [683] derived a constraint $M > 10^{-19}$ eV. The stellar heating can also lead to thicker galactic disks, possibly improving the agreement with observations for fuzzy DM with $M \sim 10^{-22}$ eV [683], which is, however, in some tension with the other bounds discussed in this section.

○ One expects **fewer sub-halos**, since 'quantum pressure' can favour their disruption beyond the usual effects of tidal forces. Comparing with, e.g., the census of MW satellites, this imposes $M > 2.9 \ 10^{-21}$ eV.

○ **Reduced dynamical friction** is expected. As discussed in section 2.5, a massive object moving through DM attracts DM, which then tends to form a slightly over-dense DM trail behind the object. The gravitational pull on the object from the over-density in the trail creates a friction-like force. This effect of dynamical friction is reduced by fuzziness, possibly by up to an order of magnitude, especially in systems with low DM velocities, because then the de Broglie wave-length is larger. If DM is ultra-light ($M \sim 10^{-22}$ eV, somewhat below the lower bound in eq. (3.1)), this could help explain the survival of globular clusters outside galaxies.

○ Fuzzy DM could form soliton-like objects, known as '**bose stars**', with mass $m$ and size $R$ such that the 'quantum pressure' balances the gravitational pull, $Gm/R \sim 1/M^2R^2$, giving for the bose star radius, $R \sim M_{\rm Pl}^2/mM^2$ (see section 10.5.4 for further details). If $m \gtrsim M_{\rm Pl}^2/M \sim M_\odot(10^{-10}\,{\rm eV}/M)$ the system collapses into a black hole, since then $R_{\rm Sch} \sim Gm \gtrsim R$. Self-interactions modify the picture: a negative scalar quartic coupling $\lambda_4 < 0$ (see eq. (3.14)) causes an instability, resulting in a collapse, $R \to 0$, such that relativistic effects become important.

In summary, while above effects do set bounds on fuzzy DM, subject to astrophysical uncertainties, they only restrict the lower end of the possible mass spectrum: $M \gtrsim 3 \ 10^{-21}$ eV, if the Eridanus II cluster disruption bound is neglected, $M \gtrsim 10^{-19}$ eV, if it is included. In addition, black hole super-radiance, discussed below, can probe a wide range of larger fuzzy DM masses.

A wider parameter space opens up, if one relaxes the requirement that the fuzzy substance constitutes all of DM. Bounds that apply in this case are summarized in fig. 6.17. Let us briefly discuss the main ones [684]. The CMB peaks are affected in various ways by the modified expansion history implied by the presence of a substance like fuzzy DM, e.g., via the suppression of small scales or an enhanced integrated Sachs-Wolfe effect on large scales. This allows to constrain the abundance of fuzzy DM to below the percent level over the range $10^{-33}$ eV $\lesssim M \lesssim 10^{-24}$ eV. The bounds become slightly more stringent, if modifications to the matter power spectrum as measured in Boss are included. Combining the CMB with the weak lensing shear measurements by Des (section 1.2.1) extends the bound to $M > 10^{-23}$ eV, if fuzzy DM has a large abundance. The galactic rotation curves of the Sparc compilation can be tested for the presence of solitonic cores predicted by fuzzy DM, implying a bound on $\rho_{\rm DM}/\rho_{\rm cr}$ at the 5% level over the range $10^{-24}$ eV $\lesssim M \lesssim 10^{-20}$ eV (partially covered in fig. 6.17), although the precise contours depend on the assumptions. The suppression of the small scale structures that is induced by fuzzy DM has the additional effect of suppressing the formation of early galaxies and early stars at high redshifts, and therefore of delaying the onset of reionization. This excludes a fuzzy component with $M \lesssim 10^{-22}$ eV from contributing more than half of DM abundance, although the bound may relax a bit following more recent determinations of the optical depth.

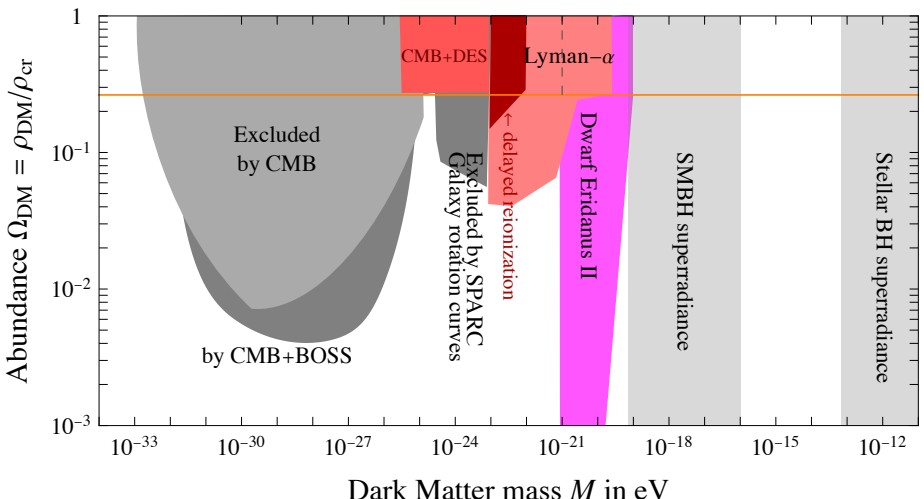

Figure 6.17: *Bounds on the* **cosmological abundance of an ultra-light (fuzzy) DM component**. *Figure adapted from fig. 3 of Antypas et al. (2022) [108]. References are listed in [684] and [683]. The figure somewhat naturally connects to fig. 10.1, where the astrophysical, cosmological and laboratory constraints on axions and axion-like Dark Matter are presented. The vertical dashed grey line signals the minimal DM mass of eq. (3.1). Laboratory constraints on ultra-light DM are discussed in section 5.8.*

Finally, let us turn to the constraints due to **black hole super-radiance**. The constraint results from the observation that light scalars would cluster around compact objects such as black holes, if their Compton wavelength is comparable to the size of the black hole. They would form a characteristic atom-like object consisting of a BH 'nucleus' and a surrounding 'cloud' of ultralight bosons. The cloud builds up and then, if the BH is spinning, starts to extract energy and angular momentum from the BH, in a phenomenon known as *super-radiance*. The measurements of BH spins can therefore be used to constrain the existence of ultra-light fields, whether they are the DM or not. Due to the required matching between the Compton wavelength of the ultra-light field and the size of the BH, black holes of different sizes probe different masses of ultra-light scalars: the super massive black holes in the centers of the galaxies probe masses in the range $7\ 10^{-20}$ eV $\lesssim M \lesssim 10^{-16}$ eV, while stellar black holes probe the mass range $7\ 10^{-14}$ eV $\lesssim M \lesssim 2\ 10^{-11}$ eV, see fig. 6.17. The bounds do depend significantly on the modeling of the process and also on the properties of the ultra-light scalars. In particular, self-interactions in the field potential can allow to evade the bounds. The implications of BH super-radiance for axion parameters are discussed in section 10.4.5.

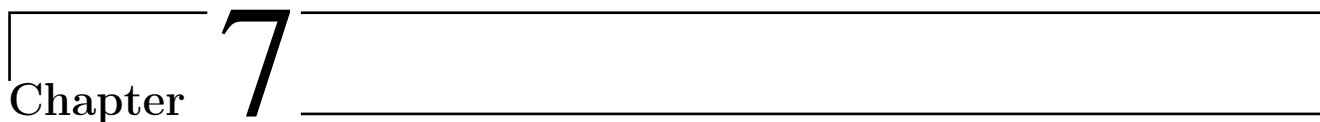

# Chapter 7

# Detection at colliders and accelerators

Production of DM at accelerators represents the third possible way in which DM can be searched for and its properties ultimately tested, see fig. 4.6. The most straightforward interpretation of DM as a thermal relic leads to a testable prediction given in eq. (4.10): a DM with order-unity couplings (as is the case for weak gauge interactions) has a TeV-scale mass, while DM is even lighter, if the couplings are smaller. This can be tested at particle accelerators – current and planned colliders are capable of producing particles in this mass range. Even if one allows larger, non-perturbative, couplings, the thermal relic DM must still be lighter than 100 TeV, see eq. (4.26), and can thus still be a target of possible future colliders.

However, even if colliders might have already produced DM particles, this by itself is not enough for a discovery. Once produced, the DM particle must also be detected, which is not completely straightforward. The only unavoidable signature of DM is missing energy (missing transverse energy at hadron colliders) since DM interacts only feebly with matter and escapes the detector, carrying with it energy and momentum. This is an indirect and rather difficult signature, and can be limited by our understanding of the backgrounds.

Specific DM models can also predict signals that are much easier to detect, at colliders or at other accelerator-based experiments. DM may be accompanied by other states, for instance new mediators, that can decay to the SM particles, giving visible imprints in the detectors. Such signatures are model-dependent, with a vast literature devoted to them. To date, no statistically significant deviations from the SM have been observed, either in missing energy or in visible channels. Conversely, a positive signal at colliders would allow to devise dedicated measurements to better understand DM using the controlled laboratory environment.

This chapter starts with a focus on 'traditional' TeV DM searches at colliders, and then enlarges its scope in later sections. After a quick overview of the basic principles of collider physics in section 7.1, we discuss the main signatures of DM at colliders in section 7.2, and compare it with other detection strategies in section 7.3. The searches for more general DM models are covered in section 7.4. These are mostly accelerator-based, but also cover particle physics experiments in a wider sense and thus even include atomic physics and astrophysics searches when these can probe the same physics.

## 7.1   A brief summary of collider physics

High energy physics is strongly affected by what can and cannot be done with collider experiments. We therefore first briefly summarize the main facts about colliders.

## 7.1.1   Production

There are two main types of colliders:

**Electron-positron colliders**. In order to reach high energies, $e^{\pm}$ are accelerated in circular orbits with radius $R$. The energy of a circular collider is limited by the energy losses via Larmor radiation, $dE/dx = 2\alpha\gamma^4\beta^3/3R^2$ where $\gamma = E/m_e$ is the Lorentz factor. Imposing that this equals the maximal energy loss, $dE/dx|_{\text{max}}$, which one is able to deliver to $e^{\pm}$ in a given accelerator using electric fields, fixes the maximal energy of an $e^+e^-$ collider:

$$\frac{E_{\text{max}}}{m_e} = \gamma_{\text{max}} \approx \sqrt[4]{R^2 \cdot dE/dx|_{\text{max}}} \sim 6 \; 10^5 \sqrt{\frac{R}{5\,\text{km}}} \sqrt[4]{\frac{dE/dx|_{\text{max}}}{m_e c^2/10\,\text{cm}}}. \tag{7.1}$$

The last such collider was LEP, that reached center-of-mass energy $\sqrt{s} \approx 200\,\text{GeV}$ and had a radius $R \approx 4.3\,\text{km}$. There are now discussions to build the next version of such a collider with $R \approx 15\,\text{km}$ and thus a higher energy. However, since the maximal energy increases only slowly with the circular collider parameters, $R$ and $dE/dx|_{\text{max}}$, the alternative is to build linear $e^+e^-$ colliders. Their maximal energy is limited by their length $L$ as

$$E_{\text{max}} = L \left.\frac{dE}{dx}\right|_{\text{max}} = 1\,\text{TeV} \frac{L}{10\,\text{km}} \frac{dE/dx|_{\text{max}}}{1\,\text{MeV}/\text{cm}}. \tag{7.2}$$

Attempts to reduce $L$ by devising technologies that allow much higher values of $dE/dx|_{\text{max}}$ are being studied. Circular muon colliders, if feasible in the future (muons need to be collimated and accelerated before they decay), would reach much higher energies bypassing the energy loss issue thanks to $m_\mu \gg m_e$.

**Hadron colliders** collide protons with protons or with anti-protons (and possibly heavier nuclei with other nuclei). The $p\bar{p}$ option leads to a better use of the collision energy, while the $pp$ option has the advantage that the proton beams can be made more intense. Since $m_p \gg m_e$ the Larmor radiation is negligible in current and past hadron colliders. Their energy is instead limited because, according to the Lorentz force, a strong magnetic field $B$ is needed to circulate $p$ or $\bar{p}$ with high energy $E$ along orbits with radius $R$ (as needed to slowly accelerate them via electric fields):

$$E = QRB = 15\,\text{TeV}\frac{Q}{e}\frac{R}{5\text{km}}\frac{B}{10\,\text{Tesla}}, \tag{7.3}$$

where we neglected the mass of $p$ or $\bar{p}$ compared to $E$. Magnetic fields exceeding tens of Tesla surpass the atomic electric fields responsible for holding ordinary matter together. Consequently, their magnetic pressure $\wp \sim B^2/2\mu_0$ is so strong that it causes mechanical devices to break apart. Furthermore, the highest magnetic fields are obtained by relying on super-conducting materials carrying large currents. Superconductivity is achieved at low temperatures, and thus requires sophisticated cryogenic systems, lots of energy, and leads to significant time waste whenever a problem arises.

The LHC $pp$ collider started in 2010 and reached $\sqrt{s} = 13.6\,\text{TeV}$ in 2022. So far its CMS and ATLAS experiments accumulated an integrated luminosity of about $100 - 200/\text{fb}$. The plan for LHC is to enter a high luminosity phase, without raising the collision energy, until $\sim 3000/\text{fb}$ are accumulated around 2035. A future $pp$ collider with $\sqrt{s} \sim 100\,\text{TeV}$ is being discussed. Assuming the cost is not prohibitively high, it will be built only many decades in the future.

While hadron colliders reach higher energies than electron colliders, their physics reach is limited by the fact that protons are not elementary particles. This implies that: a) Elementary quarks

and gluons inside protons only contain a variable fraction $x \sim 0.1$ of the total proton momentum, such that each elementary collision has a reduced energy $\hat{s} = x_1 x_2 s$. b) The large cross-section $\sigma(pp) \sim 1/m_\pi^2$, where $m_\pi = 0.14$ GeV is the pion mass, produces large QCD backgrounds to the elementary processes that one would like to investigate, which have much smaller cross sections, $\hat{\sigma} \sim 1/\hat{s}$ . Since the SM backgrounds also contain neutrinos, which are invisible to the LHC experiments just like the DM is, this presents a challenge when looking for missing energy signals.

**Fixed target experiments.** Fixed target experiments are experimentally simpler than the hadron and electron colliders, since one needs to only create one beam, which then collides with the target at rest. Due to its relative simplicity the beam can be made out of protons or electrons, but also out of muons. Another benefit is that beams can be optimised for intensity, so that one can collect large data samples of collisions. On the negative side, the center of mass of the collision, $\sqrt{s} = \sqrt{2Em_p}$, is much smaller than the energy of the beam $E$ (here $m_p$ is the mass of the proton as a stand-in for the mass of the nucleons in the target material, and we neglected the mass of the particles in the beam). Especially interesting for dark sector searches are the so called *beam dump* experiments, in which the beam is dumped into a dense block of a heavy material, so that the hadronic cascade is absorbed as fast as possible.

Since cross sections for the production of heavy new particles get smaller at higher collision energy, $\hat{s}$, intense and collimated beams are needed to see a statistically significant number of high-energy events. So far, a relatively large number of SM processes were seen at the highest energies at the LHC, with no significant excesses. This is signaling that DM is either heavier or that it has couplings small enough that it is not being produced in any appreciable quantity.

## 7.1.2   Detection

A typical particle detector only detects directly the lightest few particles, which are either stable or long-lived on collider timescales, such that they leave visible imprints:

○ **Electrons and photons** are seen in electro-magnetic calorimeters. These devices can reliably identify them and measure their energies with $\approx 1\%$ level precision.

○ **Muons** appear as almost-stable particles. A magnetic field in the external volume of the apparatus allows to identify them, distinguishing between $\mu^+$ and $\mu^-$. Additionally, their momentum can be measured from the curvature of their trajectories, provided that their energy is not much larger than 1 TeV.

○ **Quarks and gluons** appear, in collisions with energy release well above the QCD scale, as jets of hadrons whose momenta are measured in a dedicated hadronic calorimeter. In conjunction with other instruments, this allows for the identification of the main hadrons: $\pi^\pm$, $p$, $n$, $K$,... By summing their energies, the jet energy can be reconstructed with $\approx 10\%$ uncertainty. The phase space distributions of the hadrons offer only limited means to discriminate quarks from gluons. A tracker positioned near the production point can identify bottom quarks based on their displaced decays.

○ **Neutrinos** are inferred indirectly from the fact that they carry away some fraction of the total energy of the event or, at hadronic colliders, some 'transverse energy' (momentum in the direction transverse to the colliding beams).

Heavier particles can be identified via their decay products. For example, a $Z$ boson decaying into a $\mu^+\mu^-$ pair can be recognised by using their four-momenta $P_{\mu^\pm}$ to calculate the invariant mass $M_{\text{inv}}^2 \equiv (P_{\mu^+} + P_{\mu^-})^2$ and checking that it agrees with $M_Z^2$. More complicated versions of this basic idea allow

to devise kinematical variables [685] useful for partially discriminating the missing energy produced by light particles (such as neutrinos) from the missing energy produced by heavier particles (such as possibly DM).

## 7.2  Dark Matter signals at colliders

Given that DM must be a stable neutral particle, it cannot be singly produced at colliders but rather only in pairs such as DM+$\overline{\text{DM}}$ or DM+DM, possibly together with extra SM particles. More complicated DM theories may require three or more DM particles to be produced simultaneously. The collider energy therefore needs to be at least $\sqrt{\hat{s}} > 2M$, in order to be able to produce DM. Once produced, free DM particles escape without interacting with the detectors, resulting in a missing energy signal in the same way as neutrinos do.[1] Such a minimal signal of DM is detectable only if some other visible particle is also produced during the collision, otherwise the event goes unnoticed. The visible particles might be produced automatically, e.g., when DM pairs are produced inside decay chains such as in eq. (7.4). Otherwise, one needs to search for rarer events where a photon or a jet is produced together with DM (the so called *mono-photon* and *mono-jet* signatures).

### 7.2.1  Supersymmetric Dark Matter

The Higgs mass hierarchy problem motivated the presence of a new supersymmetric partner for each SM particle, with masses below about 1 TeV. The lightest of such supersymmetric particles (LSP, most probably a neutralino, here denoted as $N$ and elsewhere often as $\chi_0$) is a good DM candidate (see section 10.1.2). This supersymmetric DM scenario was so popular that the word 'neutralino' was often used as a synonym for DM, see, e.g., old reviews in [1]. Within such a scenario, $pp$ collisions were expected to produce large numbers of supersymmetric particles with strong interactions, gluinos, $\tilde{g}$, and squarks, $\tilde{q}$. These particles would ultimately decay into DM, possibly through long chains of successive decays, such as

$$pp \to \tilde{g}\tilde{g}, \qquad \tilde{g} \to \bar{q}\tilde{q} \to \bar{q}q\chi \to \bar{q}q\ell\bar{\nu}N, \tag{7.4}$$

where $\chi$ denotes a chargino and $\ell$ a charged lepton. The decay modes depend on the unknown mass ordering of sparticles. This scenario predicts a large cross section for missing energy events accompanied by possibly many leptons and jets. Many studies focused on special models (such as 'CMSSM' - the Constrained Minimal Supersymmetric Standard Model) and tried to develop general techniques for reconstructing the DM mass and the sparticle masses from invariant masses, end-points and other kinematical features of the rich decay chains.

Long decay chains, such as the one in eq. (7.4), would have been such a distinct experimental signature that already a single type of events might have enabled the simultaneous discovery of DM, supersymmetry, and Higgs naturalness (section 10.1.2). However, physicists are still waiting for statistically significant excesses of such events. Nothing like this was observed at the LHC with $\sqrt{s} = 13\,\text{TeV}$ after about $2 \times 140/\text{fb}$ of integrated luminosity.

Several other DM signals have been proposed based on different variants of supersymmetric DM. One such possibility is that the LSP is not stable but decays into gravitino DM (the gravitino is the spin-3/2 supersymmetric partner of the graviton). If the LSP is electromagnetically charged and/or carries QCD

---

[1]Alternatively, two DM particles produced almost at rest can form a dark bound state, which then decays through the annihilation of its components into visible states. This only happens in models where DM interacts via light enough particle such that dark bound states can exist. In these models, scatterings among SM particles can receive a detectable correction from diagrams where DM appears in loops [686].

color, and its decay rate is sufficiently slow, it would be possible to detect both the LSP charged tracks and the secondary vertices from the LSP decays.

Another possibility is that the LSP has such a long lifetime that it first stops in the detector and then decays only at a much later time, potentially even after the collider has already been turned off. Which decay channels dominate depends on the identity of the LSP: a photino would decay into a photon and a gravitino; a slepton would decay into a lepton and a gravitino, etc.

In light of negative results from the LHC, one hypothesis that has been entertained is that the only light sparticles are the stop $\tilde{t}$ (relevant for the Higgs mass hierarchy problem) and the DM. The expected DM signal would be the $pp \to \tilde{t}\tilde{t}^*$ production, potentially accompanied by initial-state radiation, followed by $\tilde{t}$ decays into DM, e.g., $\tilde{t} \to tN$ if $M_{\tilde{t}} > M + M_t$. The correct cosmological DM abundance is obtained for $M_{\tilde{t}} - M \approx 35\,\text{GeV}$, i.e., in the regime where stop co-annihilations play a pivotal role to explain DM as a thermal relic (see section 7.2.2 below).

## 7.2.2 Colored co-annihilations

A notable DM scenario, especially relevant for DM searches at hadron colliders, is DM accompanied by a dark sector that contains a slightly heavier colored particle, DM$'$ [687]. For example, in supersymmetric models DM is the lightest neutralino, with mass $M$, while a gluino or a squark can be only slightly heavier, with a mass $M' = M + \Delta M$. In this case, even if DM + DM annihilations are negligible, the desired DM thermal relic abundance can be obtained from co-annihilations, see section 4.1.6. The excess DM abundance is annihilated away since DM can be converted to DM$'$ through scattering on the SM plasma, after which DM$'$ annihilate through sizable cross sections dictated by QCD,

$$
\sigma(\text{DM}'\text{DM}' \to gg + q\bar{q})v_{\text{rel}} = \frac{\pi\alpha_{\text{s}}^2}{M_{\text{DM}'}^2} \times \begin{cases} 7/27 & \text{if DM}' \text{ is a scalar color triplet,} \\ 27/16 & \text{if DM}' \text{ is a scalar color octet,} \\ 7/54 + 2/3 & \text{if DM}' \text{ is a fermion color triplet,} \\ 27/32 + 9/8 & \text{if DM}' \text{ is a fermion color octet.} \end{cases} \tag{7.5}
$$

Using the results of section 4.1.6, but specializing to the case of co-annihilations in which only the colored particles can annihilate, the total DM abundance $Y = g_{\text{DM}}Y_{\text{DM}} + g_{\text{DM}'}Y_{\text{DM}'}$ is obtained from the effective annihilation cross section

$$
\sigma_{\text{eff}} = \sigma(\text{DM}'\text{DM}' \to \text{SM}) \times R^2, \qquad R = \frac{g_{\text{DM}'}Y_{\text{DM}'}^{\text{eq}}}{g_{\text{DM}}Y_{\text{DM}}^{\text{eq}} + g_{\text{DM}'}Y_{\text{DM}'}^{\text{eq}}} = \left[1 + \frac{g_{\text{DM}}}{g_{\text{DM}'}}\frac{\exp(\Delta M/T)}{(1 + \Delta M/M)^{3/2}}\right]^{-1}. \tag{7.6}
$$

The Sommerfeld and bound-state corrections, discussed in section 4.1.5, are non-negligible, as they are controlled by the strong coupling evaluated at the renormalization scale of the order of the initial-state kinetic energy. The correct thermal relic abundance is obtained for specific values of $M$ and $M'$, for instance $\Delta M \lesssim 35\,\text{GeV}$ (200 GeV) and $M \lesssim$ few TeV (10 TeV) for a fermionic color triplet (octet) [687]. From a theoretical point of view, such near degeneracy of states with differing quantum numbers would ideally require some sort of justification, but it could also be just accidental. From an experimental point of view, the cross section for production of colored particles at the LHC is much larger than it is for the direct production of DM, since the former is governed by strong interactions. Yet, the near degeneracy makes the experimental searches more challenging. Since only a small amount of collisional energy is converted into visible energy, resulting for instance in soft leptons or quark jets (for instance, due to $\tilde{q} \to q + \text{DM}$ decay), while most of the energy is carried away in the form of DM mass.

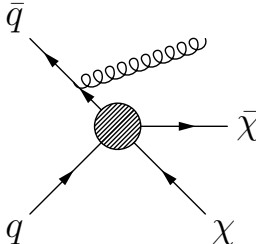

Figure 7.1: *Representative diagram for DM pair $(\chi, \bar{\chi})$ production via a **non-renormalizable operator**, represented by a shaded blob. For the process to be observable an initial-state emission of SM particles is needed. In the above diagram a gluon is emitted from the initial anti-quark leg, resulting in a mono-jet signature.*

## 7.2.3  Effective operators?

As already mentioned, no new physics beyond the SM has been found at the LHC so far. This may indicate that new physics is too heavy to be produced at the LHC (the other option is that new physics is light enough, but too weakly coupled to be produced in large enough quantities to be observed). At low energies the effects of new physics can be systematically captured by constructing appropriate effective field theories (EFTs), where the heavy mediators between DM and the SM particles are integrated out, giving rise to effective non-renormalizable operators. DM is then pair-produced via the non-renormalizable higher dimensional operators, see fig. 7.1.

Keeping all possible operators up to a particular operator dimension then captures the effects of many DM theories. This approach is systematic in its nature. One simply assumes the spin of DM (usually 0, 1/2 or 1) and writes down all possible operators up to, let's say, dimension-7 operators involving two DM particles and the SM particles [688]. An example of a dimension 6 operator is

$$\mathscr{L}_{\mathrm{EFT}} \supset -\frac{1}{\Lambda^2}(\bar{q}\gamma_\mu q)J^\mu_{\mathrm{DM}}, \qquad J^\mu_{\mathrm{DM}} \sim \begin{cases} S^* i \overset{\leftrightarrow}{\partial}_\mu S & \text{if DM is a scalar } S, \\ \bar{\chi}\gamma_\mu \chi & \text{if DM is a fermion } \chi, \end{cases} \tag{7.7}$$

where $q$ is a quark field: either the left-handed electroweak doublet $Q = (u, d)_L$ or the right handed weak singlets $u_R$ or $d_R$. The quark flavor indices are omitted. The operator is suppressed by the effective new-physics scale $\Lambda$. Full electroweak quark multiplets are needed for consistency, otherwise weak corrections grow with energy. Significant couplings to light quarks or gluons are most important for having a large cross section for DM production at the $pp$ LHC collider.

The operators in eq. (7.7) can arise from a tree level exchange of a heavy vector, $V^\mu$, with the interaction Lagrangian $\mathscr{L}_{\mathrm{int}} \supset \sum_q g_q(\bar{q}\gamma_\mu q)V^\mu + g_{\mathrm{DM}} J_\mu V^\mu$, in which case

$$\frac{1}{\Lambda^2} = \frac{g_q g_{\mathrm{DM}}}{m^2}, \tag{7.8}$$

where $m$ is the mass of the mediator and $g_{q(\mathrm{DM})}$ its coupling to quarks (DM). In deriving this expression the propagator of the mediator was approximated as $1/(q^2 - m^2) \to -1/m^2$, where $q$ is the momentum flowing through the propagator line connecting quarks to DM. This approximation is equivalent to ignoring dynamics associated with the mediator: the mediator is integrated out, rendering the interaction point-like. All the information about the mediator is still in principle available at low energies, encoded in the form of higher derivative operators (i.e., operators suppressed by powers of $q^2$) and the values of their coefficients, however the expansion is usually truncated to make the problem tractable. The EFT approximation breaks down when $q \sim m$, i.e., when the energies in the collision are comparable to the

mass of the mediator.

The EFT approach is ideal for describing the results of direct detection experiments, where $q \lesssim 200$ MeV, which is much less than the mediator mass in most DM models, see section 5.1.9. The use of EFTs to describe the results of LHC searches for DM is more questionable. The cross sections induced by non-renormalizable operators tend to grow with energy. For example, dimension-6 operators like the one in eq. (7.7) give $\sigma \sim E^2/\Lambda^4$ for the DM production cross section at colliders and $\sigma \sim M^2/\Lambda^4$ for the DM annihilation cross section. This energy dependence might suggest that the high energy hadron colliders could set stringent bounds on the EFT operators. However, the EFT interpretation faces two problems. Firstly, the highly energetic events are less probable, since they require one quark in the proton to carry a large fraction of the proton momentum. The available statistics is thus smaller, despite the larger cross sections. Secondly, the EFT description is valid only for

$$E \lesssim \min(m, 4\pi\Lambda), \tag{7.9}$$

where $m$ is the mediator mass. For energies above $\sim m$ the full theory is needed. For energies above $\sim 4\pi\Lambda$ the cross section mediated by an effective operator violates unitarity, signaling a breakdown of the theoretical description. This happens when the couplings in the full theory become nonperturbative, typically when $g \sim 4\pi$ is reached.[2] Moreover, translating the bound on $\Lambda$ into a bound on $m$ requires the knowledge of coupling constants $g_q, g_{\text{DM}}$, see eq.(7.8). The largest mass of $m$ the measurement is sensitive to is obtained in the non-perturbative limit, $g_{q,\text{DM}} \sim 4\pi$.

Typically, the bounds from the LHC searches for DM do not reach the $\Lambda \gg E$ regime, i.e., the regime for which the effective operator description is valid, except for when $g_q, g_{\text{DM}}$ are close to the non-perturbative limit. This makes the EFT approach less applicable to the interpretation of the LHC searches. Part of the problem can be traced to the fact that the typical DM signals require an emission of a visible SM particle either from the initial or the final state, resulting in the mono-jet, mono-photon, mono-$Z$, mono-higgs, ..., signals. Experimentalists need this visible particle to tag the event. The price for adding an extra particle is the reduction in the cross section by a factor $\sim g^2/(4\pi)^2 \sim 10^{-3}$, where $g \sim 1$ is the corresponding SM coupling, i.e., either the strong, the weak or the electro-magnetic coupling. As discussed above, the colliders have accumulated enough luminosity so that one is able to measure the main high-energy SM processes, but not enough to probe the processes with substantially smaller cross sections. Furthermore, at the LHC the signals of the type 'mono-something plus missing energy' are heavily affected by the SM backgrounds. Sometimes the EFT regime can be recovered by not considering the highest energy bins in the data, thus achieving that $E \lesssim \Lambda$, even though the highest energy bins are nominally the ones most sensitive to the higher dimension operators. Not using all the available data is less than ideal. This situation can be avoided by employing the simplified models, a strategy that we discuss next.

## 7.2.4   Simplified models

The problem with the effective operators can be rephrased in more physical terms: the main signal at the hadron colliders is not 'DM plus mono-something', but rather the on-shell production of new particles that mediate interactions between DM and the SM. Effective operators, introduced as an attempt to avoid dealing with the plethora of possible DM theories, do not capture the on-shell production of the mediators and thus miss the most important production channels.

It is then better to use a different approach to describe the physics of DM and the mediators at the LHC and build representative cases of DM models — the 'simplified models'. In constructing the

---

[2] The unitarity bounds computed at tree-level, with all the order one factors included, are generally not more precise than this estimate. In the large coupling regime the non-perturbative effects become important, and thus the estimates based on perturbation theory become unreliable.

simplified models we keep from the complete UV model the smallest number of new physics fields that still describes well the relevant physics: the thermal DM relic abundance and the experimental signatures at the LHC [689, 690]. Simplified models for DM typically have a mediator and the DM field, with different choices for the couplings and the spins of DM and the mediator. The full theory in most cases contains more fields, not just the DM and the mediator. These additional new physics fields are assumed to be much heavier and as such not important for the DM phenomenology.

Both in the simplified models and in the full models, DM is pair-produced via tree level exchanges of the mediator, either in the $s$- or $t$-channel, or via loop level mediator exchanges, see fig. 7.2. An example of an $s$-channel mediator is the heavy vector mediator, $V^\mu$, that we introduced in the discussion of EFTs, section 7.2.3, eq. (7.8). The most commonly considered simplified models are:

▶ **$s$-channel color-neutral vector or axial-vector mediator**. The interaction Lagrangians are (assuming flavor conservation)

$$\mathscr{L}_{\text{(axial-)vector}} = -g_{\text{DM}} V_\mu J^\mu_{\text{DM}} - \sum_f g_f V^\mu \bar{f} \gamma_\mu (\gamma_5) f, \qquad (7.10)$$

with the DM current $J^\mu_{\text{DM}}$ given in (7.7). The sum runs over the SM quarks $q = u, d, s, c, b, t$, charged leptons, $e, \mu, \tau$, and neutrinos (for which $f \to P_L \nu$ in (7.10)). To reduce the number of parameters flavor universality is usually assumed, so that the above simplified models depend only on a few parameters: the mediator mass, $m$, the DM mass, $M$, couplings to quarks, $g_q$, to leptons, $g_\ell$, and to DM, $g_{\text{DM}}$, (where $g_q \equiv g_u = g_d = g_s = g_c = g_b = g_t$, and $g_\ell = g_e = g_\mu = g_\tau = g_{\nu_e} = g_{\nu_\mu} = g_{\nu_\tau}$). In the numerical benchmarks the limit $g_{\text{DM}} \gg g_{q,\ell}$ is taken, working either in the leptophobic, $g_\ell = 0$, or the leptophilic, $g_\ell \neq 0$, limit. These choices simplify the analysis of data, but still showcase the sensitivities of various LHC searches. In concrete UV models flavor universality may not be realized, so that once DM is discovered the above assumptions can and should be relaxed.

▶ **$s$-channel color-neutral scalar or pseudo-scalar mediator**. In this case some care is required in the construction of the simplified models. For instance, considering a simplified model with a pseudo-scalar mediator $a$, the interaction Lagrangian ($y_f$ are the SM Yukawa couplings)

$$\mathscr{L}_{\text{pseudo-scalar}} = -g_{\text{DM}} a \bar{\chi} i \gamma_5 \chi - \sum_f g_f y_f a \bar{f} i \gamma_5 f, \qquad (7.11)$$

does not lead to a theoretically consistent simplified model above the electroweak scale. The interactions in (7.11) lead to unitarity violating predictions for cross sections since the Lagrangian is not invariant under electroweak gauge transformations (the scalar interactions couple SM fermions of left- and right-handed chirality, which are in different EW representations). For instance, the amplitudes $\mathscr{A}(qb \to q'ta) \propto \sqrt{s}$ and $\mathscr{A}(gg \to Za) \propto \ln^2 s$ diverge in the limit of large center-of-mass energy $\sqrt{s}$. To unitarize the two cross sections $a$ needs to be embedded in a full EW multiplet. The additional diagrams from the exchanges of a charged Higgs, $H^\pm$, which would be part of such an EW multiplet, would make the cross section predictions finite. However, the embedding of $a$ into an EW multiplet is not unique. In principle, there are many viable simplified models for $s$-channel color neutral scalar or pseudo-scalar mediators. For a pseudo-scalar mediator the community has endorsed the use of the simplest choice: a two Higgs doublet with a singlet pseudo-scalar that couples to DM (the 2HDM+$a$ model) [690]. However, it is important to remember that this is by far not the only viable possibility and that the details of LHC exclusions depend on the details of the model, such as the relative importance of the mono-jet ($pp \to j + E^{\text{miss}}_T$), mono-Higgs ($pp \to H + E^{\text{miss}}_T + X$), mono-$Z$ ($pp \to Z + E^{\text{miss}}_T + X$), $pp \to t\bar{t} + E^{\text{miss}}_T$, and $pp \to tX + E^{\text{miss}}_T$ channels.

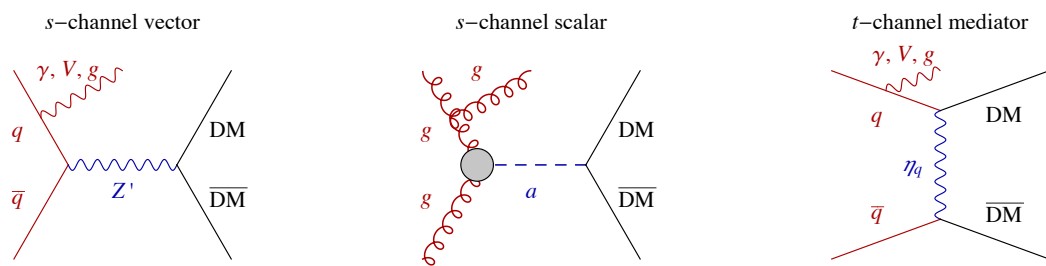

Figure 7.2:   *Representative diagrams for the DM production via s-channel (axial-)vector mediator (left panel), s-channel (pseudo-)scalar mediator (middle), and t-channel mediator (right).*

▶ **t-channel color-triplet scalar or pseudo-scalar mediator**. If the mediator, $\eta_q$, carries a QCD charge the pair of DM particles is produced via the $t$-channel exchange of the mediator. The color triplet mediator, see fig. 7.2 (right), then plays a similar role as do squarks in supersymmetry. However, in the simplified model the couplings to quarks are not constrained by supersymmetry, and can be varied freely. The interaction Lagrangian is (in Weyl notation)

$$\mathscr{L}_{\text{triplet}} \supset y_q \, q\chi\eta_q + \text{h.c.,} \tag{7.12}$$

where DM $\chi$ was taken to be fermionic, and the mediator bosonic; the other option is also viable. The EW quantum numbers of $\eta_q$ are the same as the quarks it couples to, i.e., the $\eta_q$ is an EW doublet if it couples to the left-handed SM quarks. A more common choice is that $\eta_q$ is an $\text{SU}(2)_L$ singlet that couples to right-handed SM quarks. For LHC phenomenology the flavor composition of the couplings to quarks is also important — common choices are assuming no flavor violation and either flavor universal couplings to the first two generations of quarks or a nonzero coupling to either $b_R$ or $t_R$.

There are several important differences between the phenomenology of simplified models and the EFT discussion in section 7.2.3. The comparison crucially depends on the size of the mass of the mediator, $m$, relative to the typical momentum exchanged in the collision, $\hat{s}_{\text{th}}$ (the cross sections drop quickly as $1/\hat{s}^n$, so that they are dominated by the threshold, $\hat{s}_{\text{th}}$). The $\hat{s}_{\text{th}}$ is set by the cuts imposed in the experimental analyses, such as the minimal required $p_T$ of the recoiling jet, or the amount of the missing momentum $\not{E}_T$ (momentum carried away by the DM pair). As a rule of thumb we can take, $\hat{s}_{\text{th}} \simeq 4M^2 + \not{E}_T^2$, with $M$ the DM mass. A typical analysis would vary $\not{E}_T$ between several 100 GeV up to several TeV.

Focusing on $s$-channel mediator models we can distinguish three regimes: $i)$ light mediators, $m^2 \ll \hat{s}_{\text{th}}$, $ii)$ the resonant region, $m^2 \gtrsim \hat{s}_{\text{th}}$, and $iii)$ the EFT regime, $m^2 \gg \hat{s}_{\text{th}}$. In the light mediator regime, $m^2 \ll \hat{s}$, the mediators are produced off-shell. The propagator of the mediator can be approximated by $1/(\hat{s} - m^2) \to 1/\hat{s}$. This is the exact opposite of the EFT limit, where in the mediator propagator the $\hat{s}$ can be neglected compared to the heavy mass, $1/(\hat{s} - m^2) \to -1/m^2$. Using EFT predictions in the regime of light mediators is thus not warranted and leads to erroneous constraints that are too strong (since $1/m^2 \gg 1/\hat{s}_{\text{th}}$).

In the resonant regime, $m^2 \gtrsim \hat{s}_{\text{th}}$, the mediators are produced on-shell. The $pp \to$ mediator $\to$ DM DM cross section is enhanced by the resonant production of the mediator to $\sigma(pp \to \text{mediator}) \sim g_q^2/(4\pi m^2)$, compared with the cross section $\sigma(pp \to \text{DM DM}) \sim g_q^2 g_{\text{DM}}^2 \hat{s}_{\text{th}}/((4\pi)^3 m^4)$ that is obtained when EFT is assumed to be valid. The EFT prediction for $\sigma(pp \to \text{DM DM})$ is kinematically suppressed, as well as suppressed by the relative phase space factor, $1/(4\pi)^2$, due to the emission of two DM particles instead of a production of a single mediator. The cross section calculated using the EFT therefore underestimates the real production cross section. Assuming EFT instead of the on-shell mediator production

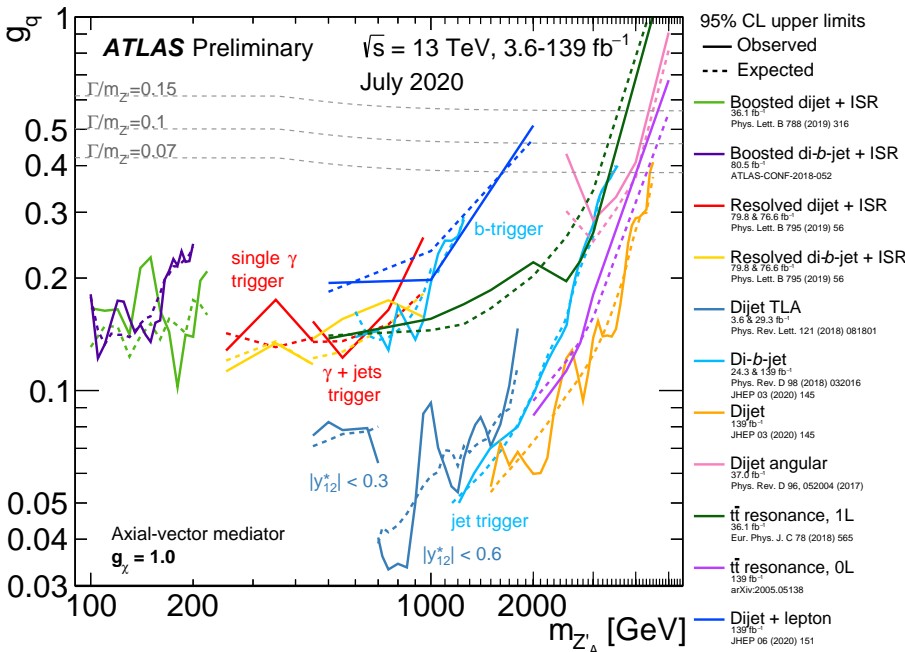

Figure 7.3: ***Constraints on axial-vector mediator*** $Z'_\mu$, *with flavor universal couplings,* $\mathscr{L}_{\text{int}} = g_q \sum_{q=u,d,s,c,b,t} Z'_\mu \bar{q} \gamma^\mu \gamma_5 q + g_{\text{DM}} Z'_\mu \bar{\chi} \gamma^\mu \chi$ *obtained by the ATLAS collaboration from various analyses of pp collisions at the LHC. In the figure the coupling to DM is fixed to* $g_{\text{DM}} = 1$ *(denoted as* $g_\chi$ *above). Figure from [691].*

thus leads to over-conservative limits in the resonant region.

In the narrow-width approximation the resonant production cross section is given by

$$\sigma(pp \to \text{DM\,DM}) \approx \sigma(pp \to \text{mediator})\text{BR}(\text{mediator} \to \text{DM\,DM}). \qquad (7.13)$$

The strength of the DM signal in the resonant region therefore depends on the branching ratio for the decay of the mediator to a pair of DM particles. For a mediator that is much heavier than the DM and the SM quarks, we have $\text{BR}(\text{mediator} \to \text{DM\,DM}) \simeq g_{\text{DM}}^2/(g_{\text{DM}}^2 + 3N_f g_q^2)$, assuming that DM is a Dirac fermion, and that only decays to $N_f$ flavors of SM quarks and to DM are open ($g_q$ is taken to be flavor universal for simplicity). For $g_q \gg g_{\text{DM}}$ the mediator decays dominantly to SM quarks, while for $g_{\text{DM}} \gg g_q$ the mediator decays invisibly to two DM particles. Other decay channels can also be open. For instance, the mediator can decay to charged leptons: the $e^+e^-, \mu^+\mu^-, \tau^+\tau^-$ pairs, or have more complicated decay chains. To properly explore the allowed parameter space of the simplified model all these final states need to be taken into account.

Finally, for heavy mediators, $m^2 \gtrsim \hat{s}$, the EFT and the simplified model descriptions coincide. The exact point where the EFT description suffices depends on the experimental errors – for less precise measurements the EFT description can be assumed already for lower mediator masses. For LHC searches the transition from the resonance region to the EFT regime typically occurs for $s$-channel mediator masses $m \gtrsim \mathcal{O}(3 \text{ TeV})$.

As we saw above, the mediator phenomenology depends crucially on the mediator mass $m$. For instance, the deviations from the effective operator limit are important if $E \gtrsim m$ and are captured by the simplified models. The other important aspect of simplified models, that is completely missed by the EFT description of DM–SM interactions, is that the mediators can be searched for using other probes, not only through the decays to DM. The mediators in general couple to quarks, leptons and DM. The

coupling to quarks and leptons can then lead to deviations in channels that have in principle nothing to do with DM. The correlations between such different probes are kept in simplified models, but are not kept in the EFT description. For instance, in EFT the four-quark operators and the quark-quark–DM-DM operators are taken to be completely uncorrelated, while in a simplified $V^\mu$ model the two corresponding coefficients are given by $g_q g_{\mathrm{DM}}/m^2$ and $g_q^2/m^2$, see eq. (7.8), and are clearly correlated.

For LHC searches to be relevant, the mediator necessarily couples to quarks and/or gluons. This means that the mediator will also decay to quarks and gluons, and the searches at the LHC such as $pp \to V^\mu \to q\bar{q}$ also apply. The creation of a quark-antiquark pair appears as two jets in the detector. The relevant experimental searches at the LHC are the bump-hunting in the dijet samples of different quark flavors. The exclusions obtained by ATLAS are shown in figure 7.3 for axial-vector (with the constraints roughly the same for vector mediators $V^\mu$). Since the dijet searches are more constraining than the missing energy searches, the null results at the LHC imply $g_q \ll g_{\mathrm{DM}}$ for simplified models that are not excluded by the LHC searches (for mediator masses, $m \in [0.1, 4]$ TeV, explored by the LHC).

As already stressed above, when constructing simplified models it is important that the models obey the electroweak gauge symmetry, since the energies involved can be well above the electroweak symmetry breaking scale. For instance, a spin 0 mediator $S$ can have Yukawa couplings to pairs of SM fermions only if it is appropriately charged under the SM gauge interactions (e.g., $S$ can be a Higgs doublet). The other option is that spin-0 mediators have trilinear or quartic couplings to the Higgs, $\mathscr{L}_{\mathrm{int}} \supset \mu S H H^\dagger + \lambda S S H H^\dagger$, and then couple to SM fermions through Higgs-$S$ mixing after Higgs obtains the vacuum expectation value, $H = (0, v + h/\sqrt{2})$. In this case $S$ can also be a singlet under the SM gauge group. The models become more complicated for the case of a pseudo-scalar mediator, see the discussion surrounding eq. (7.11).

A particularly interesting possibility for the LHC searches are the $t$-channel mediators $\eta_q$ with squark-like Yukawa couplings to quarks and DM, eq. (7.12). Since the $t$-channel mediators carry color, they can be efficiently pair produced in the $pp$ collisions, $pp \to \eta_q \eta_q^*$, and searched for through their various decays.

In conclusion, simplified models capture the relevant features of full DM models, if a mass gap separates DM and a single mediator from the rest of the spectrum. By no means does this exhaust all the possibilities, especially if the full DM theory contains a number of states that are close in mass to the DM. An important example of the latter is DM whose relic abundance is fixed by co-annihilations. In simplified models one still needs to make a number of choices for mediator couplings (for instance, the flavor composition of couplings to quarks, whether or not the mediators couple to leptons, etc). While simplified models assume minimal field content, the freedom of choice for various couplings results in simplified models that are not so very simple.

An even more minimal possibility is that the mediator is not a new physics state but rather one of the neutral SM particles, such as the Higgs or perhaps the $Z$ boson [692]. A sensitive collider signal arises if DM is so light that the SM particle can decay into DM, acquiring an invisible decay width. Bounds on the invisible Higgs branching ratio, $\mathrm{BR}_{\mathrm{inv}} \lesssim 0.11$ at 95% C.L., can be obtained either simply from the Higgs production rate (knowing that $\Gamma(X \to h) = \Gamma(h \to X)$, see e.g. [693]) or through dedicated searches [694]. These place severe constraints on the Higgs portal DM, such that DM can be the thermal relic only if it is heavy enough, so that the Higgs cannot decay into it, $M \gtrsim M_h/2$.

## 7.3  Comparing collider with direct and indirect signals

Typically, DM models lead to potential signals in many of the searches: both in direct detection, indirect detection and collider experiments (the latter in appreciable numbers only if the mediators couple to quarks and gluons). The question thus naturally arises of how to compare the possible signals.

Simplified models can be used to gauge the sensitivities of different experiments for sample DM models. As can be seen from table 7.1 the strength of the signal can vary widely, depending on the

| | Interaction | Direct detection | Indirect detection | Colliders |
|---|---|---|---|---|
| Type | $\mathscr{L}_{\text{int}}$ | NR limit, $\chi A \to \chi A$ | $\chi\bar{\chi} \to q\bar{q}$ | $\sigma(q\bar{q} \to \chi\bar{\chi})$ |
| $V \times V$ | $\left(g_{\text{DM}}\bar{\chi}\gamma^\mu\chi + g_q\bar{q}\gamma^\mu q\right)V_\mu$ | $1_\chi 1_N$ | $s$-wave | $\sigma_0$ |
| $V \times A$ | $\left(g_{\text{DM}}\bar{\chi}\gamma^\mu\chi + g_q\bar{q}\gamma^\mu\gamma_5 q\right)V_\mu$ | $1_\chi(\boldsymbol{S}_N \cdot \boldsymbol{v}_\perp), \boldsymbol{S}_\chi \cdot (\boldsymbol{q} \times \boldsymbol{S}_N)/m_N$ | $s$-wave | $\sigma_0$ |
| $A \times V$ | $\left(g_{\text{DM}}\bar{\chi}\gamma^\mu\gamma_5\chi + g_q\bar{q}\gamma^\mu q\right)V_\mu$ | $(\boldsymbol{S}_\chi \cdot \boldsymbol{v}_\perp)1_N, \boldsymbol{S}_\chi \cdot (\boldsymbol{q} \times \boldsymbol{S}_N)/m_N$ | $p$-wave | $\sigma_0$ |
| $A \times A$ | $\left(g_{\text{DM}}\bar{\chi}\gamma^\mu\gamma_5\chi + g_q\bar{q}\gamma^\mu\gamma_5 q\right)V_\mu$ | $\boldsymbol{S}_\chi \cdot \boldsymbol{S}_N, (\boldsymbol{S}_\chi \cdot \boldsymbol{q})(\boldsymbol{S}_N \cdot \boldsymbol{q})/m_\pi^2$ | $p$-wave | $\sigma_0$ |
| $S \times S$ | $\left(g_{\text{DM}}\bar{\chi}\chi + g_q y_q \bar{q}q\right)a$ | $1_\chi 1_N$ | $p$-wave | $3y_q^2\sigma_0/4$ |
| $S \times P$ | $\left(g_{\text{DM}}\bar{\chi}\chi + g_q y_q \bar{q}i\gamma_5 q\right)a$ | $1_\chi(\boldsymbol{S}_N \cdot \boldsymbol{q})m_N/m_\pi^2$ | $p$-wave | $3y_q^2\sigma_0/4$ |
| $P \times S$ | $\left(g_{\text{DM}}\bar{\chi}i\gamma_5\chi + g_q y_q \bar{q}q\right)a$ | $(\boldsymbol{S}_\chi \cdot \boldsymbol{q})1_N/M$ | $s$-wave | $3y_q^2\sigma_0/4$ |
| $P \times P$ | $\left(g_{\text{DM}}\bar{\chi}i\gamma_5\chi + g_q y_q \bar{q}i\gamma_5 q\right)a$ | $(\boldsymbol{S}_\chi \cdot \boldsymbol{q})(\boldsymbol{S}_N \cdot \boldsymbol{q})m_N/Mm_\pi^2$ | $s$-wave | $3y_q^2\sigma_0/4$ |

Table 7.1: **Comparison of different DM probes** *assuming as mediators a vector $V_\mu$ or scalar $a$, and as DM a Dirac fermion $\chi$ with mass $M$. The first two columns describe the possible chiral structures of couplings to DM and SM quarks $q$, where $y_q$ is the SM Yukawa; for notation see eq.s* (7.10), (7.11). *The third column lists the leading non-relativistic operators giving rise to DM scattering on nuclei. The fourth column indicates whether the $\chi\bar{\chi} \to q\bar{q}$ annihilation proceeds through s-wave or p-wave. The last column gives the total cross section for $q\bar{q} \to \bar{\chi}\chi$ in the limit where the center of mass energy, $\sqrt{s}$, is much larger than the DM mass, but still much smaller than the mediator mass. The partonic cross section $\sigma_0 = g_{\text{DM}}^2 g_q^2(s/m_{\text{med}}^2)^2/(6\pi s)$ needs to be convoluted with the parton distribution functions in order to obtain the $pp \to \chi\chi$ cross section, relevant for the LHC.*

detailed structure of the mediator couplings. In particular, for direct detection and indirect detection the chiral structure of the DM and SM currents is very important. This is a consequence of the non-relativistic nature of DM in the initial state in both direct and indirect detection. Since the axial and vector (similarly, scalar and pseudo-scalar) currents have drastically different non-relativistic limits, this translates into very different expected signals. In table 7.1 we list a few illustrative examples: the four chiral structures of currents for either vector or scalar mediators. The expected sizes of direct and indirect detection signals for many other DM/SM couplings can be found in [695].

The third column in table 7.1 lists the leading DM/nucleon non-relativistic operators that describe the scattering in direct detection experiments. These operators still need to be sandwiched inside nuclear wave functions to obtain the rate for DM scattering on nucleus (for details see section 5.1). However, already at the level of DM/nucleon interactions, and ignoring the complications introduced by the nuclear physics, it is clear that taking the non-relativistic limit leads to drastically different kinematical suppressions for various chiral structures. For instance, the $V \times V$ and $S \times S$ interactions give rise to the numbering operators in the non-relativistic limit (the operator $1_\chi$ in table 7.1 counts the number of DM particles, while $1_N$ counts the number of nucleons). Consequently, the spin-independent DM/nucleus scattering is coherently enhanced, schematically $\sigma \propto A^2$, where $A$ is the mass number of the nucleus, since the nuclear matrix element is proportional to the number of neutrons and protons in the nucleus. Such enhancement is not present for the other chiral structures. The $A \times A$ interaction leads to spin-dependent scattering (in table 7.1 $\boldsymbol{S}_\chi$ denotes the DM spin, $\boldsymbol{S}_N$ the nucleon spin), while the remaining chiral structures are even further kinematically suppressed by either the momentum exchange, $\boldsymbol{q}$, or by the initial relative velocity between DM and the nucleon, $\boldsymbol{v}_\perp$ (a scaling discussion of relative sizes of different contributions can be found in [696]).

In indirect detection only the DM is non-relativistic, while the outgoing quarks are highly relativistic (we assume $M \gg m_q$). The relative angular momentum between $\chi$ and $\bar{\chi}$ determines the velocity suppression of the $\bar{\chi}\chi \to q\bar{q}$ annihilation cross section, $\sigma v_{\text{rel}} = \sigma_L v_{\text{rel}}^{2L}$, see also the discussion below eq. (4.16). That is, the $L = 0$ (s-wave) annihilation is not velocity suppressed, while $L = 1$ (p-wave) annihilation is. The transformations of the DM bilinear under rotations, combined with $C$ and $P$ transformations, fully determine the total angular momentum, the spin and the orbital angular momentum of the initial

state that this bilinear annihilates, and thus whether the *s*-wave annihilation is possible. We see that the pattern of suppressed and unsuppressed signals is quite different from the one in direct detection. For instance, while in direct detection both $V \times V$ and $S \times S$ interactions lead to coherently enhanced DM/nucleus scattering, in indirect detection the $S \times S$ interactions leads to velocity suppressed *p*-wave annihilation, in contrast to $V \times V$, which gives rise to the unsuppressed *s*-wave annihilation. Similarly, while $V \times A$, $P \times S$, and $P \times P$ lead to suppressed direct detection signals, they give rise to unsuppressed *s*-wave annihilation signals in indirect detection.

The final column in table 7.1 gives the expected sizes of the DM pair production cross sections at colliders such as the LHC. Instead of discussing the production cross sections for the monojet signal, $pp \to \chi\bar{\chi} + \text{jet}$, we use as a proxy the size of the simpler $q\bar{q} \to \chi\bar{\chi}$ scattering cross section. Even so, the conclusions we draw apply more generally. For simplicity we work in the limit of heavy mediators and with the energy of the collision much greater than the DM mass. Unlike the direct and indirect detection, for collider physics the chiral structure of the interactions has little effect on the size of the production cross sections. This is easy to understand from naive dimensional analysis, since the only scale in the problem is the collision energy, $\sqrt{s}$. The structure of the interaction only tells us whether just left-handed or right-handed or both chiralities participate in the interaction, which changes the predictions for the cross section by factors of 2. In the examples in table 7.1 the cross sections for the vector mediator (and separately for the scalar mediator) do not even differ by such factors of 2, and are instead the same for all chirality structures, a consequence of the fact that the interactions have definite transformation properties under parity.

In conclusion, the observation of a signal in one set of experiments and non-observation in another could pinpoint to the nature of DM interactions with the SM.

# 7.4   Searches with beam dumps and precision measurements

Dark matter could be part of a more complicated dark sector. This has significant consequences for experimental searches. Instead of just searching for the DM particle, a more general search strategy tries to produce any of the (possibly lighter) particles in the dark sector. The discovery of such mediators between visible and dark sector may then give access to the DM itself as well. In such 'fishing expeditions' the theoretical bias is best kept to a minimum. However, completely ignoring theoretical inputs would leave us with a gigantic space of possibilities. A more useful approach is to, at least initially, focus on interactions with the lowest dimension, the so called *portals*. Keeping only renormalizable operators and operators suppressed by a single power of the high scale gives four distinct possibilities for the mediators:

- ○ **a light scalar,** usually dubbed either a *"dark Higgs"* or a *"light singlet"*, that mixes with the SM Higgs through the portal interaction;

- ○ **a light pseudo-scalar**, usually called *"axion"*, with derivative couplings to the SM fermion currents, suppressed by a high scale $f_a$;

- ○ **a spin 1/2 fermion**, dubbed *"heavy neutral lepton"* or *"sterile neutrino"* $\nu_s$, that couples to the SM leptons and the Higgs via a Yukawa interaction;

- ○ **a spin 1 vector boson**, dubbed *"dark photon"* or *"light Z′"* and denoted as $A'$, that couples to the SM via kinetic mixing.

The theoretical aspects of dark sector portals are discussed in more detail in section 9.4, with a summary given in table 9.3. Here, we mostly concentrate on the experimental searches for the dark sector mediators, at accelerators or more generally in laboratory set-ups.

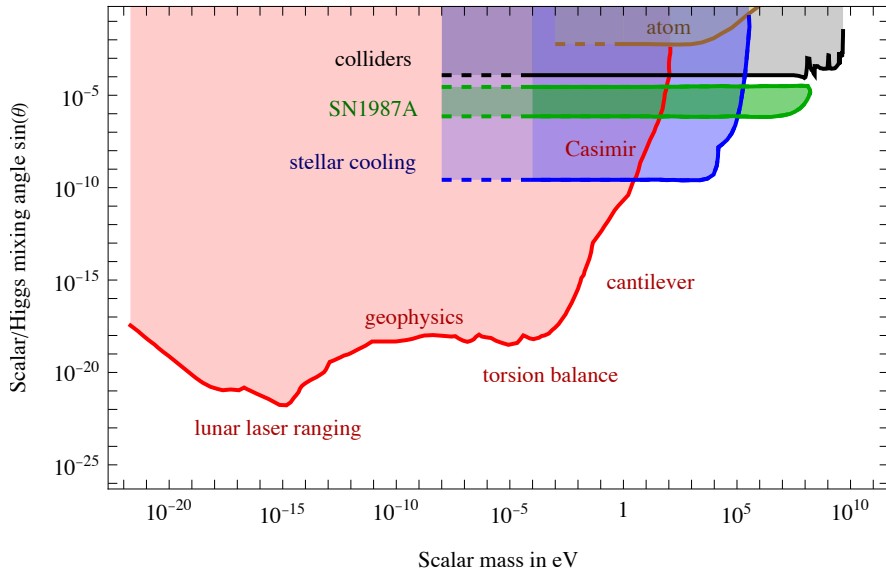

Figure 7.4: ***Constraints on a light scalar that mixes with the Higgs*** *as a function of the mixing angle θ for a wide range of scalar masses [697–702]. In the constraints we assumed that the coupling to DM is subdominant to the Higgs-induced couplings to the SM fermions. If the coupling to DM is important, this would affect only the high-mass boundaries of the collider (black), supernova (green) and stellar cooling (blue) exclusions. There are further constraints from colliders for scalar masses in the 100 GeV to TeV range, which are not shown, since they are model dependent (they do not depend just on the mixing angle).*

The mediators to dark sectors, listed in table 9.3, can be produced directly in the laboratory experiments, or observed indirectly through effects due to virtual exchanges of the mediators. The phenomenology depends crucially on the mass of the mediator, and falls roughly in one of the three regimes. For mediators lighter than an eV, the Compton wavelength is of atomic size or larger, all the way to macroscopic or astrophysical sizes. For masses in the MeV to TeV range the mediators can be produced in typical particle physics experiments. Heavier mediators, in contrast, can only be observed through their off-shell exchanges. Light particles are subject to strong bounds that forbid electric, weak and strong charges, in agreement with DM properties.

## 7.4.1   Mediators with sub-eV masses

Very light mediators, with sub-eV masses, generate new macroscopic forces. A tree level exchange of a light $Z'$ vector mediator or a light scalar of mass $m_{\rm med}$ generates a potential between two particles of the form

$$V_{ij}(r) = -\alpha_{ij} \frac{e^{-m_{\rm med}r}}{r}. \tag{7.14}$$

The coupling constant is $\alpha_{ij} = +g_i g_j/4\pi$ for the scalar mediator, and $\alpha_{ij} = -g_i g_j/4\pi$ for the vector mediator, where $g_i$ is the coupling constant for interaction between the mediator and the $i$-th particle. This means that the potential between two particles of the same specie — for instance, between two protons or two electrons — is always attractive (repulsive) for the scalar (vector) mediator. For particles of different species, e.g., between a proton and an electron, the potential can be of either sign. Furthermore, the form of the potential in eq. (7.14) is not the only possibility: for instance, a pseudo-scalar mediator or a mediator that couples to the SM fermions through higher dimension operators would induce spin-

dependent potentials, see, e.g., [703].

The range of the potential is controlled by the mass of the mediator, and is given by

$$\lambda \equiv \frac{1}{m_{\text{med}}} = 197\,\text{nm}\,\frac{\text{eV}}{m_{\text{med}}} = 0.2\,\text{m}\,\frac{\mu\text{eV}}{m_{\text{med}}} = 1.3\,\text{a.u.}\,\frac{10^{-18}\text{eV}}{m_{\text{med}}} = 0.6\,\text{kpc}\,\frac{10^{-26}\text{eV}}{m_{\text{med}}}, \tag{7.15}$$

where we chose numerical examples for the mediator mass that correspond to atomic, macroscopic, solar system and galactic sizes, respectively. In view of such disparate scales there are a number of very different experimental searches for mediator-induced long range potentials. Many of these searches share a strong similarity with those discussed in section 10.4.5 for axion DM.

### Fifth force experiments

Fifth force experiments measure the gravitational force between two objects using laboratory setups. They search for deviations from the $1/r^2$ dependence by comparing the force at two or more distances [698]. The limits on the strength of non-gravitational interactions is given in terms of $\alpha \equiv \alpha_{ij} G/m_i m_j$, where $G$ is the Newton constant, and $m_{i,j}$ are the masses of the two test objects. That is, for $r \ll \lambda$ the constant $\alpha$ gives the strength of the "fifth force" relative to the force of gravity. The strongest constraints are obtained for $100\,\mu\text{m} \lesssim \lambda \lesssim 10\,\text{m}$ using torsion balances, giving $\alpha \lesssim 10^{-3}$. For $\lambda \simeq 10\,\mu\text{m}$ to $\sim 100\,\mu\text{m}$ the cantilevers are used as force sensors instead of torsion balances, while for even smaller distances, up to $0.1\,\mu\text{m}$, the experiments measuring the Casimir forces are used. Since at such distances the electromagnetic interactions become important, the sensitivity quickly deteriorates all the way up to $\alpha \sim 10^{30}$. At larger distances the constraints come from the tests of Keplerian motion and have decent sensitivity up to $\lambda \gtrsim 10^{14}$ m. The best sensitivity is reached at scales $\lambda \approx 10^8$ m, where the bounds from lunar laser ranging give $\alpha \lesssim 10^{-10}$.

### Tests of the weak equivalence principle

It the coupling constant $\alpha_{ij}$ in eq. (7.14) is not proportional to the product of test masses, $m_i m_j$, then the force induced by the light mediator violates the weak equivalence principle as it depends not just on the total mass of the probe but also on the type of the material used in the experiment [698]. Tests of the weak equivalence principle are in general more sensitive than the tests of the $1/r^2$ dependence. They probe scales from cm to a.u., where the precise sensitivity dependends on the relative sizes of the couplings $g_i$ for different SM fundamental particles (the couplings to $u$ and $d$ quarks, to gluons, and to electrons).

### Atomic transitions

At smaller distances, comparable to the size of an atom, the dominant SM force is the electromagnetic interaction. The presence of portal mediators with $\lambda \sim \mathcal{O}(100\,\text{nm})$ would correct the $1/r$ electrostatic potential between the electrons and the atomic nucleus, changing the atomic levels away from the SM predictions. The experimental precision on these is stunning, with the fractional frequency precision of atomic frequency standards based on microwave and optical transitions reaching the levels $\delta\nu/\nu \sim 10^{-16}$ and $\delta\nu/\nu \sim 10^{-18}$, respectively [699]. The main problem in converting the experimental measurement to bounds on new physics particles is that the precision of the SM theoretical predictions are nowhere near the above precision. One can bypass this problem, at least to some extent, by taking advantage of the special features of the new physics signal.

For instance, if the dark mediator has couplings that are not just the rescaled electric charges, then one can use the measurements of atomic transitions for several different isotopes to cancel out the leading electromagnetic effects. In this way one can place nontrivial bounds on the new physics contributions to the electron–nucleus potential [700].

Another interesting possibility opens up, if either the dark Higgs or the axion constitute part or the majority of the DM relic density. The occupation number per quantum state is large, $N \gg 1$, see section 3.4. Such background fields therefore behave as classical fields that oscillate coherently with the frequency equal to the mass of the mediator, $m_{\mathrm{med}}$, and induce a time dependent change in the potential between electrons and the nucleus. As a result the atomic energy levels become time dependent, a feature that can be searched for. The results are usually presented as bound on time dependence of the fundamental constants ($\alpha_{\mathrm{QED}}$, quark masses, etc.). One can also search for spatial variations in the fundamental constants, which would be induced by the spatial variations in the background dark Higgs or axion fields.

## Stellar cooling

Light dark sector particles can be copiously produced in the stars, thus modifying the stellar dynamics [701]. An outgoing flux of dark sector particles would carry away energy, possibly leading to excessive cooling. This leads to stringent constraints on the intermediate range of dark sector couplings to the visible matter: the couplings large enough such that the dark sector particles are still copiously produced in the stars, but small enough so that the dark sector particles escape from the star and are not trapped in its interior, are excluded. For dark sector particles to be produced the temperature inside the star needs to be high enough, i.e., the temperature needs to be comparable to the particle's mass. White dwarf cooling thus constrains dark sector particles with masses below a few keV, red giant cooling constrains masses below several tens of keV, and SN1987A observations below about 100 MeV.

The stellar cooling constraints are inherently uncertain, since they rely on our understanding of astrophysics. This is especially true for the bounds from SN1987A that rely on the observation of a single supernova event [702]. The exotic cooling would shorten the neutrino signal that was observed coincident with the supernova discovery in optical light, but only if the event was due to the standard core collapse supernova [704].

## Axion searches

Many laboratory experiments are searching for axions. Most of these are explicitly designed to search for the specific coupling of axion to photons, $aF\tilde{F}$, and are reviewed in section 10.4.5.

## Light shining through wall

Light shining through wall searches can be used to search for axions (see section 10.4.5) and for dark photons. Dark sector particles are created in the first part of the laser beam, before it hits the wall. The dark sector particles are the only ones that can pass through the wall, and are then converted back into a visible signal behind the wall. A version of this concept is the DARKSRF experiment, which used superconducting radio frequency cavities both for creation of dark photons (in the first cavity) and the conversion (in a different, shielded cavity) and achieved best sensitivity, below fifth force searches, for dark photon masses around $\mu$eV, excluding mixing angles larger than $\epsilon \gtrsim 10^{-8}$ [705] (see also results from the SHANHE experiment [706]).

## Virtual corrections

Some observables have been precisely measured, such as the magnetic moments of the electron and the muon, $(g-2)_e$ and $(g-2)_\mu$. Others are very small in the SM, such as the electric dipole moment of a neutron (nEDM). These could be affected by quantum corrections from DM running in the loop [707].[3]

---

[3]See he discussion of $(g-2)_\mu$ experimental anomaly and possible connection to DM in section 8.3.1, and the $m_W$ anomaly in section 8.3.4.

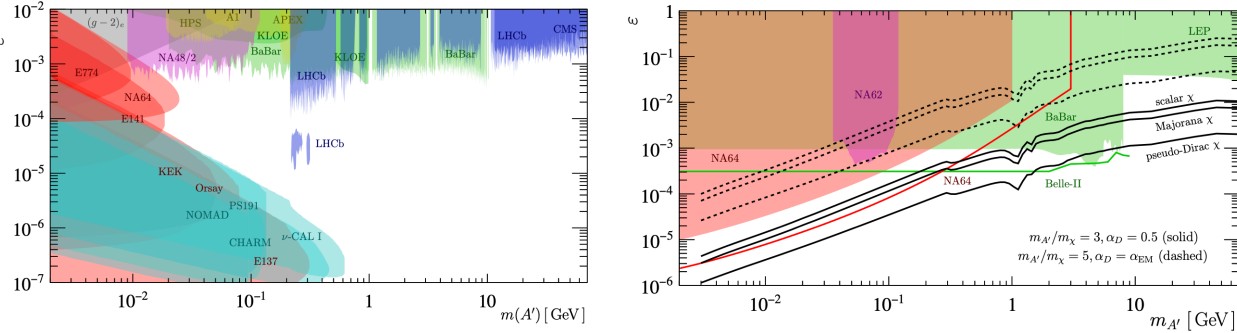

Figure 7.5: **Left**: *Constraints on a vector boson mediator (usually termed a **dark photon**, or, sometimes, a light $Z'$) in the plane of the kinetic mixing parameter $\epsilon$ (which controls the strength of interactions with the SM, see section 9.4.1) and of its mass, $M_{A'}$, assuming only the visible decays are allowed. The constrains are from electron beam dumps (dark red region), proton beam dumps (dark green), $e^+e^-$ colliders (light green), pp collisions (blue), meson decays (magenta), $(g-2)_e$ (grey), and electron on fixed target experiments (yellow). The triangular shape of the beam dump constraints is due to the fact that the $A'$ decay length scales as $1/\epsilon^2 M_{A'}$. When the $A'$ mass or the mixing $\epsilon$ are too large, the vector boson decays too fast and does not reach the detector. The collider constraints in the upper part of the plot are instead essentially flat in $M_{A'}$. These searches scan for a new resonant bump in the invariant mass distribution of two SM particles, which would indicate the presence of $A' \to SM\,SM$ decays. **Right**: sensitivity to a vector boson mediator (dark photon) that decays invisibly. The solid and dashed black lines denote several DM benchmarks for which the correct thermal relic abundance is obtained. Both figures are from Graham et al. in [710].*

While the relevance of these constraints depends on the model (for instance, nEDM requires CP violating couplings of DM), they can, in parts of the parameter space, be more stringent than the more direct searches for DM. Note that the constraints from virtual corrections are relevant both for light DM as well as for heavy DM, with mass in the TeV regime, since DM does not need to be produced on-shell.

### Neutrino oscillations to sterile neutrinos

Light fermions, with masses below eV, that are not charged under the standard model group are conventionally called sterile neutrinos, $\nu_s$. They can have the Yukawa interaction, $\mathcal{L} \supset y_s \nu_s L H$, with the SM neutrinos, which are part of the weak doublets $L$ (we are suppressing flavor indices). After electroweak symmetry breaking the Yukawa interaction, setting the Higgs field to its vacuum expectation value, $\langle H \rangle = (0, v)$, becomes a mass term, $y_s v\, \bar{\nu}_s \nu_L$, that mixes sterile and active neutrinos. Such small mass mixings can be searched for in neutrino oscillation experiments, similarly to how the SM neutrino masses themselves were discovered [708]. The interest in light sterile neutrinos has been motivated for the past two decades by a number of experimental anomalies: the LSND and MiniBooNE anomalies, the reactor anti-neutrino anomaly, and in experiments with intense radioactive sources [709]. However, none of these anomalies seems very plausible at present.

## 7.4.2   Mediators with MeV or GeV masses

This is the typical mass range where particle physics experiments can probe on-shell production of the mediators. The specifics of each search depends on the production mechanism (such as which particles are being used in the collisions) as well as on the decay products. Figure 7.5 provides a summary of the current constraints.

**Searches in $e^+e^-$ colliders**

A major benefit of the $e^+e^-$ colliders is that the events are clean, with relatively little background activity in the detector [711]. The detectors typically cover almost the complete $4\pi$ solid angle around the interaction point, apart from the small region very close to the beam pipe, which cannot be instrumented. The visible final states can therefore be fully reconstructed. This in turn means that the direction of the missing momentum and/or missing energy, due to the dark sector particles escaping the detector, can be determined precisely. The high-luminosity $e^+e^-$ colliders are primarily used for measurements of rare process with heavy quarks and are thus not very energetic. For instance, the $B$ factories (BABAR in the USA and BELLE in Japan in 2000s), and the upcoming super-$B$ factory BELLE II in Japan, have the $e^+e^-$ collision energy tuned to the resonant production of the $B\bar{B}$ meson pairs which occurs close to the production threshold $\sqrt{s} \approx \mathcal{O}(10 \text{ GeV})$. Only mediators lighter than the collision energy can be probed through on-shell production, limiting the dark sector mass reach of such experiments. More massive mediators were probed at LEP, $\sqrt{s} \approx 200$ GeV (in the future at proposed FCC at CERN or CEPC in China). The mediators can be either produced directly in the $e^+e^-$ collisions, such as $e^+e^- \to A'\gamma$, or from decays of other particles, for instance $e^+e^- \to \tau^+(\tau^- \to \nu_{\rm s}\mu^-\bar{\nu}_\mu)$. The mediators can either: *i)* decay immediately (prompt decays), *ii)* decay away from the interaction point but still in the detector, leading to displaced vertices, or *iii)* completely escape the detector leading to missing energy signature. The searches need to be optimized for each of these possibilities, as well as for the choice of the final state.

**Searches in hadron colliders**

These searches use the $pp$ collisions at the LHC, which lead to complicated final states with many pions, and other hadronic states produced in a typical collision [712]. Due to the messy hadronic environment the searches for dark sector particles at the LHC are challenging. A mitigating strategy is to look for final states and observables that are easily distinguishable, such as the resonant peak in the $\mu^+\mu^-$ and $e^+e^-$ invariant mass spectra due to $A' \to \mu^+\mu^-$ or $A' \to e^+e^-$ decays (such searches were performed or are being planned by the LHCb, CMS and ATLAS experiments). The other option is to search for mediators that are produced in the $pp$ collision, but decay far away from the collision point. The backgrounds are reduced to essentially zero by placing the detector behind a shield made of meters of iron, concrete or dirt such that the electromagnetically and strongly interacting SM particles get stopped, while dark sector particles penetrate unobstructed through the shield. Experiments of this type are FASER, and the planned experiments CODEX-B and MATHUSLA, each to be positioned close to a different collision point in the LHC ring.

**Searches in electron beam fixed target experiments**

In these types of experiments a high intensity electron beam is sent on a thin target (or a collection of such targets, most often made of high-$Z$ materials). The electrons in the beam can then produce dark photons or ALPs in the field of the nucleus in the target, while their decays are measured in detectors placed down-stream from the target. The benefit over $e^+e^-$ collisions is the potential for very high delivered luminosity, however, due to the kinematics of a collision with a particle at rest the center of mass energy is lower, and therefore the reach in the mediator mass is typically lower than in the $e^+e^-$ collisions or at the hadron colliders. Experiments of this type are A1 (Mainz Microtron), APEX (JLAB), HPS (JLAB), and NA64 (CERN) [713].

**Searches in electron and proton beam dump experiments**

In these types of experiments a high intensity electron or proton beam is sent on a thick target. The main difference with the fixed target experiments is that the beam dump experiments have detectors

behind substantial shielding that absorbs the SM particles. Such experiments are thus not sensitive to prompt decays of mediators, but only to mediators with lifetimes long enough to decay in the detector far from the beam dump target. There are a number of proposed beam dump experiments with significantly improved projected reach, including SHiP and SHADOWS at CERN, and PIP2-BD at Fermilab [714].

### Searches in meson and lepton decays

Another possible search strategy, with some overlap with the previous ones, is to search for rare decays of mesons ($\pi^0, \eta, K^+, B^0, \ldots$) or leptons ($\tau, \mu$) into dark sector + SM particles, but with the final state fully reconstructed [715]. The benefit is that this reduces possible backgrounds. In addition, weakly decaying SM particles much lighter than the weak scale have small SM decay rates, $\Gamma_{\rm SM} \sim m^5/M_W^4$ with $m \ll M_W$. This is beneficial when searching for new physics via rare decays, since the branching fraction for the rare process gets relatively enhanced by the small total decay width, $\mathrm{BR}_{\rm rare} = \Gamma_{\rm rare}/\Gamma_{\rm SM}$.

## 7.4.3   Very heavy mediators, TeV masses or above

Heavy mediators, with masses above a few TeV cannot be produced on-shell in the laboratory experiments since their mass exceeds the collision energy. Nevertheless, we can still search for them indirectly, through their virtual corrections. In order to increase the sensitivity to new physics it is advantageous to choose processes that are extremely rare in the SM, can be well predicted theoretically, and at the same time can receive large contributions from virtual exchanges of the mediators.

### Flavor-changing neutral current processes

Flavor-changing neutral currents (FCNCs), such as $\bar{b}s \leftrightarrow \bar{s}b$ or $\bar{s}d \leftrightarrow \bar{d}s$, do not occur in the SM at tree level since the neutral gauge bosons (the $Z$ boson, the gluons and the photon) only have flavor-diagonal interactions [716]. The FCNCs first occur in the SM at one loop from $W$ exchanges in the loop, and are suppressed by the loop factor and the products of CKM matrix elements and quark masses. This additional suppression is a manifestation of the "GIM mechanism" — if all the quark masses were the same there would be no flavor violation.

   The mediators can have flavor-violating couplings, in which case already the tree level exchanges of mediators induce FCNCs. Taking the corresponding flavor-violating couplings to be $\mathcal{O}(1)$, and to have $\mathcal{O}(1)$ CP-violating phases, the comparison of $K \leftrightarrow \bar{K}$ (i.e. $\bar{s}d \leftrightarrow \bar{d}s$ at quark level), $D \leftrightarrow \bar{D}$ ($\bar{u}c \leftrightarrow \bar{c}u$), $B \leftrightarrow \bar{B}$ ($\bar{b}d \leftrightarrow \bar{d}b$) and $B_s \leftrightarrow \bar{B}_s$ ($\bar{b}s \leftrightarrow \bar{s}b$) mixing rates with the SM predictions bounds vector mediators to have masses $m \gtrsim 2 \cdot 10^4$ TeV, $10^4$ TeV, $10^3$ TeV, $400$ TeV, respectively [717]. These bounds scale as $m/g_{qq'}$, and thus crucially depend on the size of the flavor violating coupling, $g_{qq'}$, that induces the $q \to q'$ transition. If $g_{qq'}$ is small, the bounds on the mediator mass $m$ become weaker: if the suppression is of the same form as in the SM, i.e., a product of CKM matrix elements and a loop factor, then mediators with weak scale masses are still allowed; if the couplings are even further suppressed, the mediators can be correspondingly lighter. Even more stringent bounds follow from the absence of FCNCs in the lepton sector, for instance, from the bounds on $\mathrm{BR}(\mu \to e\gamma), \mathrm{BR}(\tau \to 3\mu)$, etc [717].

### Electric and magnetic dipole moments

The FCNC observables are not the only precision observables that give stringent bounds on potential off-shell contributions from the mediators. For instance, electric dipole moments (EDM) are CP violating but flavor diagonal, i.e., they do not require the change of flavor. In the SM the EDMs are highly suppressed: the quark EDMs only arise at three loop order in the SM, while the electron EDM only arises at four loop order in the SM. In contrast, if the mediators have CP-violating couplings they can induce EDMs already at one loop, and thus the experimental bounds on the EDMs translate to very strict bounds

on mediator masses [718]. For instance, the bound on the electron EDM implies $m \gtrsim 6 \cdot 10^5$ TeV for mediators with maximally CP violating couplings of $\mathcal{O}(1)$. The EDM constraints of course do not apply, if the mediators have only CP-conserving couplings.

Another set of flavor diagonal but CP conserving observables that are sensitive to radiative corrections from mediators are the anomalous magnetic moments. Especially interesting is the anomalous magnetic moment of the muon, $(g-2)_\mu$, since there are indications that its measured value might deviate from the SM predictions (at $\approx 4\sigma$, if the main SM hadronic effects are calculated using dispersion relations; no significant discrepancy is seen, if the similarly accurate lattice determinations are used instead). The possible $(g-2)_\mu$ anomaly can be explained by additional contributions from the mediators running in the loop. The models are of two types: a light mediator with $m \lesssim \mathcal{O}(100 \text{ MeV})$ for which the required chirality flip comes from the external muon leg, or from $\mathcal{O}(\text{TeV})$ dark sectors where the chirality flip occurs in the mediator loop [719], see also section 8.3.1.

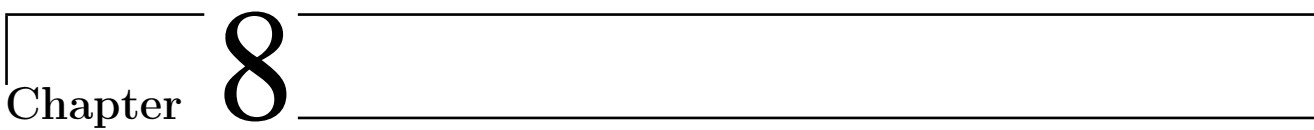

# Chapter 8

# Anomalies

In this chapter, we review the recent *anomalies*, loosely defined as departures from expected SM physics or astrophysics that may potentially involve DM as an explanation. The various anomalies are listed in table 8.1 and illustrated in fig. 8.1. Somewhat arbitrarily, we restrict the discussion to the anomalies that have attracted significant interest from the scientific community over the past 10-20 years. Although for many of them their true nature will be clarified only by future investigations, some have already been dismissed, at least in the context of possible DM interpretations (such as the 135 GeV $\gamma$-ray line in indirect detection, and the XENON anomaly in direct detection). Nevertheless, it is instructive to review these cases, as they offer insights into the practical nuances of DM searches. It remains to be seen whether any of the anomalies will persist and become the central topic of a future DM review, or if all of the anomalies will turn out to be false alarms. Some of the current anomalies may simply be due to under-appreciated systematic or theoretical uncertainties, which in many cases can only be estimated approximately. Moreover, whenever large numbers of quantities are being measured, rare statistical fluctuations are inevitably expected to occur. Research is inherently a complex endeavour, often punctuated by unsuccessful ventures.

## 8.1 Direct detection: current anomalies

### 8.1.1 The DAMA/LIBRA anomaly

The amount of target material that can be used in a particular DM direct detection experiment is limited by the stringent requirements on background reduction. A larger target mass is useful only if a smaller relative background level can be achieved, so that one can still efficiently search for an excess of DM scattering events. In late 1990s and early 2000s the DAMA/LIBRA experiment was able to use the world's largest detector mass by relaxing the background requirements and searching instead for an annual modulation in the DM direct detection signal in NaI crystals. At that time this also translated to world's leading sensitivity to spin-independent DM scattering, see fig. 5.2.

As discussed in section 5.1.7, the rotation of the Earth around the Sun causes a variation in the flux of DM particles hitting the Earth depending on the different periods of the year, such that the DM excess would peak around June 2. The DAMA collaboration claimed the observation of an annual modulation with the expected phase. The upgraded DAMA/LIBRA detector confirmed their earlier result improving the statistics and reaching a significance of roughly $13\sigma$ for the cumulative exposure [288,309], see fig. 8.2. A naive DM interpretation would need a spin-independent cross section of about $10^{-40}\,\mathrm{cm}^2$, and a fit to the energy spectrum suggests a DM mass around $10\,\mathrm{GeV}$.

For a while both the COGENT and CRESST collaborations found anomalies possibly compatible

| When | Where | | Who | What | Stat | Status |
|------|-------|------|-----|------|------|--------|
| 2025 | 8.2.14 | [720] | Km3Net | $E_\nu \sim 200\,\mathrm{PeV}$ | $2\sigma$ | astrophysics? |
| 2024 | 8.4.3 | [721] | Desi | $w_{\mathrm{DE}} \neq -1$ | $3\sigma$ | will be tested |
| 2022 | 8.2.13 | [723] | Lhaaso/Carpet-2 | 18 TeV/251 TeV $\gamma$ | ? | unclear |
| 2022 | 8.3.4 | [724] | CDF | Too large $W$ mass | $7\sigma$ | disconfirmed |
| 2022 | 8.2.2 | [725] | Lorri | Cosmic optic light excess | $4\sigma$ | disfavoured |
| 2020 | 8.1.3 | [726] | Damic | Low energy excess | $5.4\sigma$ | will be tested |
| 2020 | 8.5.4 | [98] | Pulsar Time Arrays | Stochastic gravity wave | $4\sigma$ | astrophysics? |
| 2020 | 8.1.2 | [727] | Xenon1t | $e^-$ excess recoils at 2.4 keV | $3-4\sigma$ | disconfirmed |
| 2019 | 8.5.3 | [728] | Ligo/Virgo | BH in astro mass gap | ? | will be tested |
| 2018 | 8.3.3 | [729] | Multiple experiments | $n$ decay to invisible | $4\sigma$ | will be tested |
| 2018 | 8.4.5 | [730] | Edges | 21 cm intensity | $3.8\sigma$ | will be tested |
| 2018 | 8.2.12 | [731] | Anita | Upward CR around EeV | events | to be tested |
| 2017 | 8.2.11 | [732] | Dampe | CR $e^\pm$ excess at 1.4 TeV | $2.6\sigma$ | fluctuation? |
| 2017 | 8.4.4 | [733] | Multiple surveys | CMB dipole vs distant matter | $\sim 5\sigma$ | will be tested |
| 2016 | 8.2.10 | [430] | Ams talks | CR $\overline{\mathrm{He}}$ | events | to be tested |
| 2016 | 8.3.5 | [734] | ATLAS and CMS | $\gamma\gamma$ peak at 750 GeV | $4\sigma$ | disappeared |
| 2016 | 8.4.1 | [735] | Planck vs SN | Incompatible Hubble rate | $3-6\sigma$ | to be tested |
| 2016 | 8.4.2 | [736] | $\Lambda$CDM fits vs local observations | $S_8$ matter inhomogeneity | $3\sigma$ | disconfirmed |
| 2015 | 8.2.6 | [737] | Fermi | $\gamma$-rays from the Ret II dwarf | $\sim 3\sigma$ | not confirmed |
| 2015 | 8.3.6 | [738] | Atomki | Boson at 17 MeV | $7\sigma$ | not in MEG II |
| 2015 | 8.2.9 | [739] | Pamela, Ams | CR $\bar{p}$ excess at 10-20 GeV | $\lesssim 2\sigma$ | astro? |
| 2014 | 8.3.2 | [740] | LHCb | $R_K^{(*)}$ flavour ratios | $\lesssim 5\sigma$ | disconfirmed |
| 2014 | 8.2.3 | [741] | $X$-ray telescopes | 3.5 keV $X$-rays | $4.4\sigma$ | will be tested |
| 2012 | 8.2.7 | [742] | Fermi | 135 GeV $\gamma$-ray line | $3\sigma$ | disappeared |
| 2011 | 8.5.2 | [743] | Simulations vs Observations | Too big to fail | ? | astro? |
| 2009 | 8.2.1 | [744] | Arcade-2 | Excess isotropic radio emission | $> 5\sigma$? | will be tested |
| 2009 | 8.2.5 | [745] | Fermi | GeV $\gamma$-rays from the GC | $\infty$ | astro (sources)? |
| 2008 | 8.2.8 | [166] | Pamela, Ams | CR $e^+$ around 100 GeV | $\infty$ | astro (pulsar)? |
| 2006 | 8.3.1 | [746] | Bnl E821 | Muon magnetic moment | $5\sigma$ | lattice $\approx$ SM |
| 1999 | 8.1.1 | [288] | Dama | Seasonal variation | $13\sigma$ | not confirmed |
| 199X | 8.5.1 | [747] | Simulations vs Observations | Cusp-core | ? | astro? |
| 199X | 8.5.1 | [748] | Simulations vs Observations | Diversity | ? | astro? |
| 199X | 8.5.2 | [749] | Simulations vs Observations | Missing satellites | ? | astro? |
| 1976 | 8.5.2 | [750] | Observations | Plane of satellites | ? | astro? |
| 1970 | 8.2.4 | [751] | Many | 511 keV $\gamma$-rays from the GC | $\infty$ | astro? |

Table 8.1: *Recent **anomalies** tentatively interpreted as Dark Matter or related to it.*

Astrophysics                                                      Background

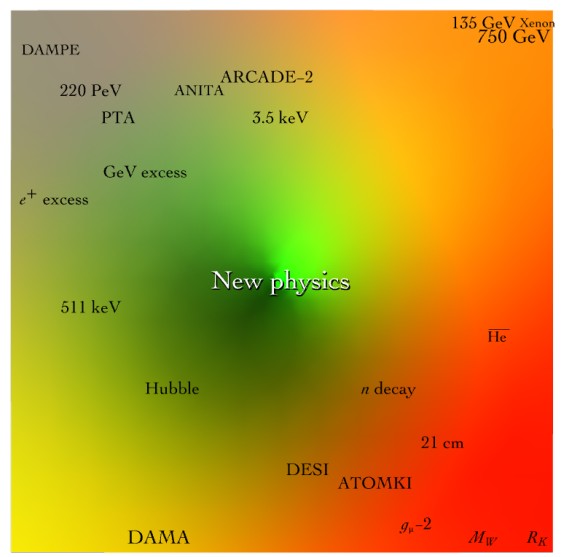

Figure 8.1: *Status map of recent **anomalies** tentatively interpreted as Dark Matter or related to it. A confirmed discovery would appear in the central green region, while anomalies possibly affected by experimental (orange), backgrounds (yellow), astrophysical issues (black), or by the lack of plausible theory (yellow) appear at the borders.*

Theory                                                            Experiment

with DAMA. Subsequent experiments, however, ruled out this particular region of the parameter space. Given the potential importance of the DAMA signal, a number of alternative models featuring additional DM properties that could enhance DAMA's sensitivity relative to the other experiments, thus evading the bounds at the time, were explored: different couplings to $p$ and $n$, inelastic DM, multi-component DM, DM scattering on electrons, etc [752]. However, as direct detection experiments advanced, they established orders of magnitude stricter bounds (see fig. 5.2 and 5.12), rendering these proposed scenarios increasingly untenable.

Although the DAMA/LIBRA anomaly seems no longer interpretable in terms of DM, the collaboration continues to support the claim of an anomaly. Various possible backgrounds and other (instrumental or environmental) effects that could account for the claim have been proposed [753], and have been refuted by DAMA, see the DAMA papers [288]. The experiments ANAIS [336] and COSINE [335], which utilize NaI(Tl) crystals akin to DAMA, are investigating the same signal. First results by ANAIS are consistent with no annual modulation in their counting rate, contradicting the DAMA/LIBRA claim at about $3\sigma$ [336]. Results from COSINE also strongly constrain the DAMA claim, disagreeing with it at more than $3\sigma$ [335].

### 8.1.2 The XENON1T anomaly at 2.4 keV

In 2020 the XENON1T collaboration reported an excess peaked at around 2.4 keV in the low-energy spectrum of electron recoil events [727]. The local statistical significance was around 3-4$\sigma$, although already at that time it was suggested that part of this excess could be due to a small tritium background in the detector. After implementing measures to mitigate tritium contamination, the successor XENONnT experiment in 2022 did not observe the previously reported excess [727].

During the interim, various dark matter-related explanations were proposed for the original observation [727]:

- The signal could have been due to new light particles created with keV energies in the Sun (the solar core has a temperature of a few keV) and then absorbed by electrons in the XENON1T detector.

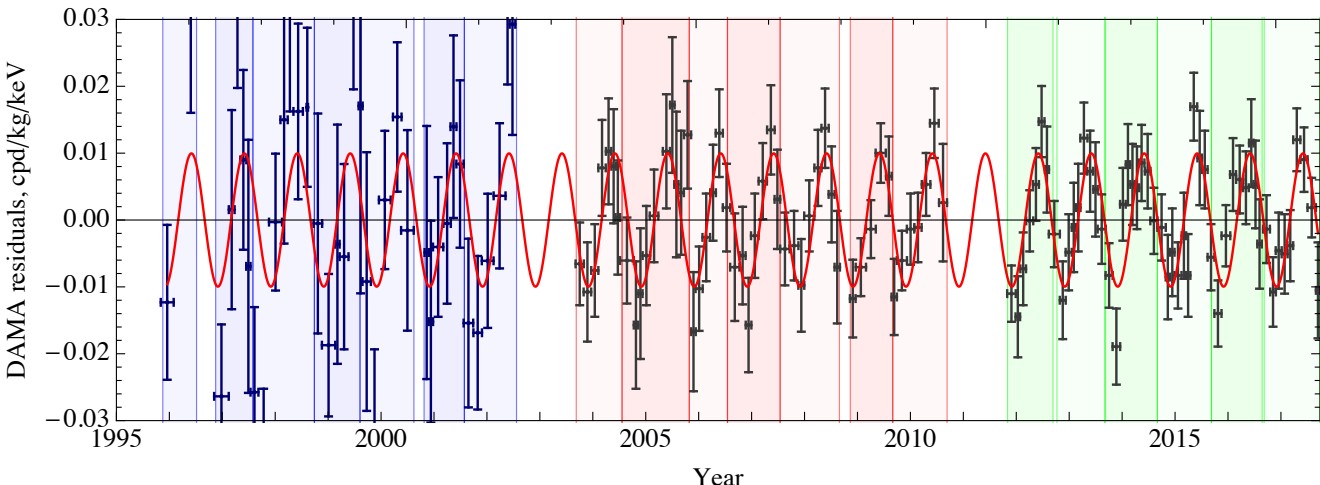

Figure 8.2: *The crosses represent* **DAMA data***: the residuals in the* 2-6 keV *recoil energy window computed by subtracting the average rates in the roughly annual cycles chosen by DAMA and plotted as vertical bands. The best fit to a Dark Matter signal (cosine annual modulation peaking at June 2) is denoted with a red curve.*

However, this interpretation was disfavoured by the bounds on the cooling rates of hotter or denser stellar systems.

○ DM as a scalar with a mass $M \approx 2.4\,\mathrm{keV}$ would produce a signal peaked at $M$, if absorbed by an electron in the detector.

○ Particle DM elastically scattering on an electron would transfer enough recoil energy, if DM is fast, with $v_{\mathrm{DM}} \sim 0.1$ (or larger if $M < m_e$). DM faster than the galactic escape velocity might be produced, e.g., by DM semi-annihilations, or by DM scattering with the Sun.

○ Inelastic scatterings of the usual slow (cold) DM on nuclei could produce a peak around 2.4 keV, if the scattering is exothermic, i.e., into an appropriately lighter state.

### 8.1.3 Low recoil energy excesses

The current direct detection experiments have reached sensitivities to sub-keV recoil energies. At energies below 1 keV, some of these experiments are observing event rates that are well above the known backgrounds [726]. The excesses were found in a number of experiments that use different target materials, types of sensors, surrounding holding structures, and are operating at different background levels. Some of these experiments are taking data on the surface, others in shallow underground facilities, and some in deep underground laboratories. It is highly unlikely that all the observed excesses are due to DM; however, some of them could be. That is, most of the observed excesses are most likely due to some not yet fully identified background source, with possible candidates as varied as: the cracking of the epoxy used to glue components of the detector, luminescence of the material in direct vicinity of the target material, transition radiation, excess neutron backgrounds, micro-fractures from the clamping of the detector, etc.

Perhaps the most intriguing is the $5.4\sigma$ excess of low-energy ionization events in the bulk of DAMIC, DAMIC-M and SENSEI skipper charge-coupled devices, housed in the DAMIC cryostat at SNOLAB, confirming the previous observation of the excess in the DAMIC detector in 2020. At present there is no

conventional instrumental explanation for the DAMIC excess. Intriguingly, a DM particle with mass and and cross section

$$M \approx (2-4)\,\text{GeV}, \qquad \sigma_p \approx \text{few } 10^{-38}\,\text{cm}^2 \tag{8.1}$$

can explain the excess events. However, this DM explanation does require isospin-violating SI couplings, such that scattering on Ar largely or completely cancels (the couplings to protons and neutrons need to have opposite signs). The explanation using isospin conserving SI couplings (the same couplings to protons and neutrons) is excluded by CDMSLite and by DarkSide-50. Similarly, DM spin-dependent scattering is excluded by CDMSLite, LUX, and PICASSO [726].

## 8.2 Indirect detection: current anomalies

### 8.2.1 The ARCADE-2 excess

In 2009, the ARCADE-2 collaboration measured the cosmic radio-wave emission over a wide range of frequencies. Above a few GHz the detected emission agrees with the expected CMB signal. For frequencies between 22 MHz and a few GHz, however, the collaboration reported a significant excess [744]. Expressed in terms of the absolute temperature of the diffuse isotropic sky (see eq. (6.52)), the emission can be fit with $T(\nu) \simeq 1.2\,\text{K}\,(\nu/\text{GHz})^{-2.60}$. The intensity of this excess emission is difficult to explain with known astrophysical mechanisms, such as the active galactic nuclei or star-forming galaxies, since it exceeds the flux predicted from astrophysical sources by a factor of approximately $5-6$.

Dark Matter has been proposed as a possible explanation [744]. DM annihilations in a large number of extragalactic halos may provide the additional source of $e^\pm$ that would emit synchrotron radiation (see section 6.7.3) and would power the emission detected by ARCADE-2. The excess is found to be well fit by a rather conventional candidate: a DM particle with a mass $M \approx 10-25\,\text{GeV}$ that annihilates into $\mu^+\mu^-$ (the measured hard synchrotron emission requires a hard $e^\pm$ spectrum, such as the one produced by a leptonic annihilation channel), with a thermal annihilation cross-section.

A possible challenge to the DM interpretation, though, is the smoothness of the observed radio signal: the spatial fluctuations are very small, possibly inconsistent with DM annihilating in clustered structured at relatively low redshift, and annihilations at high redshift might be in conflict with the EDGES limits discussed in section 8.4.5. On the other hand, baryonic matter is also clustered and thus such a smoothness is also challenging for the proposed astrophysical explanations. Alternative dark sector explanations have also been proposed, involving dark photons that oscillate into ordinary photons contributing to the radio background. Future radio-telescopes measurements, such as SKA, will be needed to clarify the nature and the origin of the excess.

### 8.2.2 The COB and CIB excesses

The cosmic optical background (COB) consists of the total integrated light that has been emitted by all stars and galaxies over the history of the Universe, and that falls in the visible band. It can be estimated on the basis of galaxy counts, performed, e.g., with the Hubble Space Telescope (HST). Its detection, however, is challenging, because of the foreground provided by zodiacal light (sunlight scattered by local dust). The NEW HORIZONS probe, a NASA mission dedicated to study the outer bodies of the solar system such as Neptune and Pluto, is in a good position to measure the COB, as zodiacal light is reduced at those locations. The Long Range Reconnaissance Imager (LORRI) instrument onboard NEW HORIZONS has reported the COB's intensity to be $16.37 \pm 1.47\,\text{nW}\,\text{m}^{-2}\,\text{sr}^{-1}$ at the pivot wavelength of $0.608\,\mu\text{m}$ [725]. This is a factor of 2 and about $4\sigma$ above the estimate from the HST galaxy counts.

Bernal et al. (2022) [725] interpreted the excess in terms of axion-like DM particles with mass $M \simeq 5-25\,\text{eV}$ that decay into photon pairs. The required coupling is $g_{a\gamma\gamma} \sim \text{few}/(10^{11}\,\text{GeV})$, which falls in the

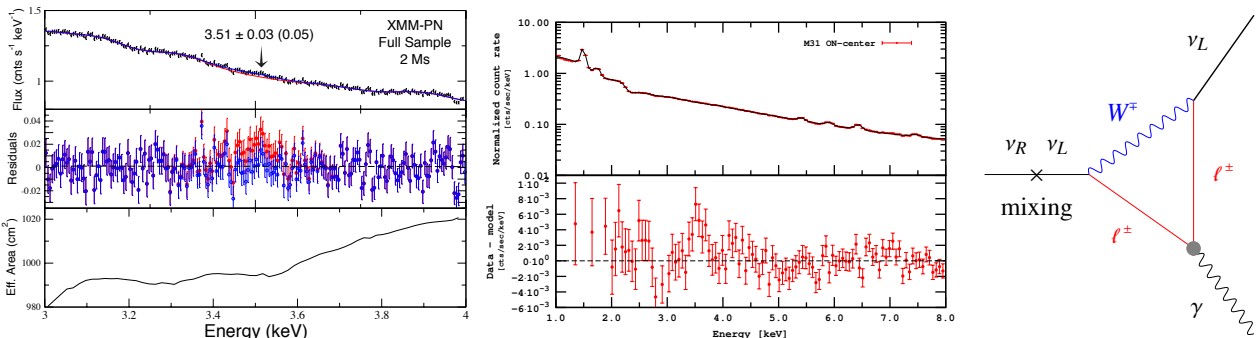

Figure 8.3: *Left and Middle:* XMM-NEWTON *X-ray data from the Perseus cluster and the Andromeda galaxy on which the* **claims of a 3.5 keV line** *are mainly based (from Bulbul et al. (2014), ©AAS, reproduced with permission, and Boyarsky et al. (2014) [741], ©APS, reproduced with permission). Right: Feynman diagram for the decay of a sterile neutrino, $\nu_R$, producing an X-ray $\gamma$ with energy equal to half of the $N_s$ mass.*

allowed window of the axion-like parameter space (axions are discussed in section 10.4: see eq. (10.36) for the definition of $g_{a\gamma\gamma}$, and fig. 10.1 for current bounds). However, this interpretation is in strong tension with the measurements of the *anisotropy* of the COB. Since the DM distribution in the Universe is clumpy, the photon signal from decaying DM is expected to not be isotropic. The predicted anisotropy is in contradiction with the upper bounds provided by the HST.

Similarly, the Cosmic Infrared Background Experiment (CIBER) reported an excess in the cosmic infrared background (CIB) radiation [754]. This measurement is affected by larger uncertainties than the COB excess, mostly due to the relevance of the zodiacal light to the measurements of CIBER, which is an Earth-based sounding rocket. The CIB excess has been interpreted in terms of $\sim 2 - 3\,\mathrm{eV}$ axion-like DM particles decaying into photon pairs, with coupling $g_{a\gamma\gamma} \sim 1.5/(10^{10}\,\mathrm{GeV})$. In this case the study of the anisotropy does not seem conclusive, but the larger coupling is in tension with stellar cooling bounds, see fig. 10.1.

### 8.2.3 The $X$-ray line at 3.5 keV

In 2014, Bulbul et al. and Boyarsky et al. [741] both reported an independent detection of an $X$-ray line at $E_\gamma \simeq 3.55\,\mathrm{keV}$, cf. fig. 8.3, using observations from XMM-NEWTON and CHANDRA of several galaxy clusters (notably, the flux from the Perseus cluster appeared to be anomalously high) and of the Andromeda galaxy M31 [741]. Subsequent works claimed the same signal, albeit with varying degrees of significance, in various other targets: the Galactic Center, patches of the Milky Way halo, other — but not all — clusters, some 'blank-sky' fields...; and with other telescopes: NuSTAR, SUZAKU. At the same time, other studies claimed no detection, again in various regions: in the GC, patches of the MW halo, dwarf galaxies including Draco, stacked galaxies, individual clusters... A reanalysis by Dessert et al. (2023) [741], for instance, questioned even the very existence of the signal for the 3.5 keV line in the data. Given the complexity of the searches, the varied nature of the targets, and dependence on the modelling of the environments and the backgrounds, it is fair to say that while the detection and no-detection claims are in strong tension, they are not yet definitively contradictory. The HITOMI experiment [517], which had the best energy resolution, did not observe the 3.5 keV line. Unfortunately, the gathered data was of very limited statistics since only slightly more than a month after launch in 2016 the satellite spun out of control and ultimately disintegrated. Nevertheless, it did report a hint for

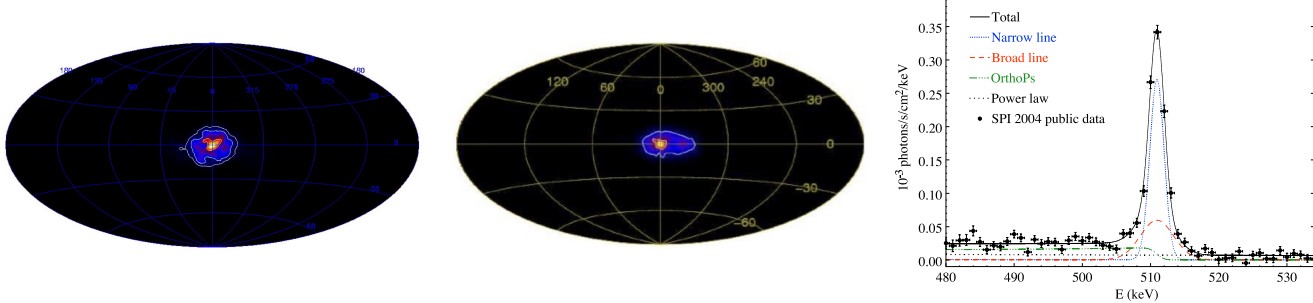

Figure 8.4: *Left: Initial **511 keV** map from* INTEGRAL-SPI *data, showing the **diffuse emission around the Galactic Center**. Middle: More detailed* 511 keV*, after 5 years of* INTEGRAL-SPI *data, from which both a spherical bulge component and a disk component can be extracted. Right: Spectral fit of the emission. The figures are, respectively, from Knödlseder et al. (2005), from the* INTEGRAL *website (see Weidenspointner et al. (2008)) and from Jean et al. (2006), in [751] (©ESO, ESA/*INTEGRAL*/MPE, reproduced with permission).*

a broadened excess at roughly the same energy.

Even if the 3.5 keV line is truly present, there is an ongoing debate whether it can simply be an unidentified atomic line emitted by excited interstellar and intergalactic gas. The intensely discussed possibilities are a potassium line and charge exchange reactions between sulfur and hydrogen.

The 3.5 keV line can be produced by $M \approx 7.1$ keV DM particle decaying into $\gamma\gamma$ or $\gamma\nu$. A signal with approximately the correct strength, would, for instance, be obtained from $\gamma\nu$ decays of sterile neutrino DM, which was produced via non-resonant oscillations with mixing angle $\sin 2\theta_{\mathrm{mix}} \approx 10^{-5}$ (section 9.2.2). Such DM, however, would be too warm and thus not compatible with the Lyman-$\alpha$ bound in eq. (3.9). Different mechanisms that produce somewhat slower DM particles are phenomenologically more viable and would also help alleviate the anomalies in galactic sub-haloes and cores (section 8.5.1).

A feature that can discriminate between a DM and a non-DM origin of the signal is that in clusters of galaxies DM is predicted to have a significant velocity dispersion. DM decays would therefore produce a broader line than atomic emissions. This is quite intriguing in light of the HITOMI hint mentioned above.[1] Other DM-related explanations include the inelastic scattering of DM particles to an excited state that subsequently decays ('excited', 'fluorescent' DM).

Future X-ray surveys will be needed to shed light on the origin and precise features of the excess, e.g., the ATHENA satellite [541], XRISM (the successor of HITOMI) [528], or the more futuristic LEM [755]. A smoking gun of DM would be the detection of this excess in regions abundant in DM and scarce in gas, e.g., in dwarf galaxies. However, current searches in these regions have not yet yielded any positive results. Another unambiguous indicator would be to detect a Doppler shift of the line, with opposite signs in the east and west hemispheres of the Galaxy. Indeed, should the line originate from DM decays in the static DM halo, the motion of the solar system would cause an average blue shift in the signal coming from the hemisphere facing the direction of motion, and a red shift in the opposite direction.[2]

## 8.2.4   The $\gamma$-ray line at 511 keV

---

[1]Since the DM speed is of the order of $10^{-3}c$, the broadening is expected to be of the order of $10^{-3}$ 3.55 keV $\simeq 3$ eV. This requires an exquisite energy resolution, which is, however, within capabilities of HITOMI and its successors.

[2]Given the typical solar speed in the galactic frame, $v_{\odot} \approx 233$ km/s, such a shift in frequency is comparable to the broadening due to the DM peculiar velocity, and within the energy resolution of forthcoming instruments.

A 511 keV emission coming from the region around the Galactic Center has been detected since the 1970's by a number of satellite and balloon missions [751]. The INTEGRAL/SPI instrument has produced the most accurate measurements to date. Recently, the COSI balloon has also detected and started characterising this emission.

The spectrum of the emission matches the expectations from $e^+e^-$ annihilations, as it peaks at $510.954 \pm 0.075$ keV, which is precisely the electron mass. It is best described (see fig. 8.4 right) as a superposition of two Gaussian components and a subdominant, lower-energy continuum. The first Gaussian is narrow (with width $\sim 1.3$ keV) and corresponds to direct $e^+e^-$ annihilation. The second Gaussian is broader (with width $\sim 5.4$ keV) and is expected when $e^+e^-$ annihilate after forming positronium in a gaseous environment. The continuum corresponds to the annihilation of ortho-positronium into $3\gamma$.

The measured flux implies the annihilation of $(2-5)\,10^{43}\,e^+$ per second, corresponding to a remarkable instantaneous luminosity of about $\sim 10^4$ times the luminosity of the Sun, or to the annihilation of $\sim 3M_\odot$ over $10^{10}$ years, assuming a steady-state condition. The morphological analysis shows the superposition of two contributions: a somewhat spherical 'bulge' and a fainter 'disk' correlated with the galactic plane. The bulge extends to about $10°$ around the galactic center, while the disk has a vertical thickness of $4° - 7°$ and a radial extent that does not go beyond $|\ell| \approx 50°$ longitude. The bulge/disk luminosity ratio is about 1.4.

The disk emission can be explained, at least in large part, by the $e^+$ injected by unstable nuclei produced in massive stars and supernovae, such as $^{26}$Al, $^{44}$Ti and $^{56}$Co. Alternative astrophysical explanations include fall-out from accretion onto the central black hole, X-ray binaries (the same ones possibly responsible for the GeV $\gamma$ excess from the GC, see section 8.2.5) and neutron star mergers. The bulge emission, however, can not be easily explained in terms of astrophysics, as it extends in latitude well beyond the region populated by stars. For this reason it has attracted a lot of attention, including explanations in terms of DM, which is naturally expected to be spherically distributed around the GC. An important constraint, both on DM models and on any other source of positrons, has been highlighted by Beacom and Yüksel (2006) [751], who pointed out that the positrons must be injected with energies $\lesssim 3$ MeV otherwise they could annihilate 'in flight' while still carrying a lot of energy, producing an unobserved higher-energy gamma ray emission spectrum.

The DM interpretations are mostly of two types: light annihilating DM or de-exciting DM. A decaying DM explanation, on the other hand, seems to be disfavored as it gives too flat of a profile, with the predicted flux falling too slowly away from the GC. In the annihilation case, DM has to be lighter than about 3 MeV, given the Beacom and Yüksel constraint, and the flux is fit with an annihilation cross section

$$\langle \sigma v \rangle_{\text{DM DM} \to e^+e^-} \approx 5 \cdot 10^{-31} \,\text{cm}^3/\text{s}. \tag{8.2}$$

These values of mass and cross section are allowed by the CMB constraints (see section 6.11.2) but would, in minimal models, lead to a DM relic density that is too large (for thermal relic DM, the observed DM abundance requires an annihilation cross section in the early Universe which is 5 orders of magnitude larger that the one in eq. (8.2), see eq. (4.13)). Nevertheless, viable models can be constructed, e.g. assuming that the cross section in eq. (8.2) is $p$-wave suppressed, which implies then a less efficient suppression in the early Universe. The Beacom and Yüksel upper limit on the DM mass can also be slightly relaxed, e.g. if the DM distribution at the GC features a pronounced density spike: this enhances annihilations, thereby requiring a cross section even smaller than eq. (8.2) and thus suppressing the 'in flight' contribution (De la Torre Luque et al. (2024) [751]).

In the second category, heavier, WIMP-like DM particles are assumed to be collisionally excited to a state split by $\lesssim 3$ MeV above the ground state. The subsequent de-excitation produces $e^+e^-$ pairs, of which the positrons subsequently annihilate with ambient electrons and produce the signal. TeV particles acquire a kinetic energy in the Galaxy of about 500 keV (given $v \sim 10^{-3}c$), which, if transferred to the excitation energy, is in the right ballpark to construct viable models.

The idea of checking the DM claim by searching for the same signal in dwarf satellite galaxies has so far given inconclusive results.

## 8.2.5   The $\gamma$-ray GeV excess from the Galactic Center

Since 2009, several authors have reported the detection of a gamma-ray excess from the inner few degrees around the GC, extending out to 10 or 20 degrees, at energies between 0.5 and 5 GeV [745]. The spectrum and the nearly spherical morphology of the emission were found initially to be compatible with the expectations from annihilating DM particles. One of the most detailed analyses is by Daylan et al. (2014) [745], which finds that the excess has a high level of significance, and is best fit by a $\sim 45\,\mathrm{GeV}$ DM particle, distributed according to a contracted ($\gamma \simeq 1.26$) NFW profile, and annihilating into $b\bar{b}$ with a cross section $\langle\sigma v\rangle = (1.4 - 2) \times 10^{-26}$ cm$^3$/s, which is compatible with the requirements for a thermal relic DM.[3] Fig. 8.5 shows both the earliest and one of the more recent fits to data. Thanks to the estimate of background modelling uncertainties quantified in Calore et al. (2014), it was soon realized that other good fits to the data are also possible, notably for DM annihilating into leptonic channels (pointing to lighter DM) and for gauge boson channels (pointing to heavier DM). The FERMI collaboration (FERMI-LAT coll. (2016)) essentially confirmed these findings, although doubts arose given that similar excesses seem to appear elsewhere in the Galactic plane (Calore et al. (2015), FERMI-LAT (2017)).

It is important to bear in mind that the identification of an 'excess' relies on the ability to carefully assess the background from which the excess is hypothesized to emerge. In the present case, the extraction of the residuals strongly relies on the modeling of the diffuse gamma-ray background (in particular the model made publicly available by the FERMI collaboration) as well as on additional modeling of astrophysical emissions. This includes emissions from FERMI bubbles, isotropic component, unresolved point sources, molecular gas,... This 'template fitting' strategy is subject to intrinsic uncertainties and prone to somewhat arbitrary modelization choices, hence it is often criticized. Nonetheless, a consensus appears to exist that an excess relative to the expected templates is present.

Before attributing the excess to DM one should consider possible alternative astrophysical explanations. A population of milli-second pulsars (MSP) has been a possibility from the very beginning. Initial detailed studies using wavelets analysis and photon-count statistics have shown that an interpretation in terms of a large number of unresolved point sources, such as MSPs, is statistically favored with respect to a diffuse emission (Bartels et al. (2015) and Lee at al. (2015)). This was later contested: Leane and Slatyer (2019, 2020) questioned the robustness of the photon-count statistics method and revived the DM hypothesis (as did List et al. (2020)), while Chang et al. (2019), Buschmann et al. (2020) and Calore et al. (2021) maintain that MSPs are preferred. Macias et al. (2016), Bartels et al. (2017), Song et al. (2024) and others, on the other hand, argued in favor of a non-DM origin of the excess mostly on the basis of its morphology, which appears not to be completely spherical, and thus better correlates with the stellar population of the galactic bulge. Cholis et al. (2022), in contrast, argued against the MSP origin of the excess, since they observed a high-energy tail in the spectrum, which MSPs cannot produce. Holst and Hooper (2024) also argued against the MSP option, claiming that FERMI should have already seen dozens of MSPs, based on the luminosity function inferred from the observations. Indeed, directly detecting these putative MSPs remains a challenge, as their expected number is highly dependent on the assumed luminosity function, whose properties are mainly inferred from the detected MSPs within the Galactic disc (not the center). Positive claims by the FERMI Collaboration (Ajello et al. (2017)) have later been corrected. Future telescopes, such as SKA, may offer more insight by detecting the associated radio emissions.

---

[3]It is important to note that the astrophysical boost factor due to DM subhalos (see section 6.8.1) is typically not included in these analyses, hence the agreement between the two annihilation cross sections could be modified. However, in this case the boost factor is expected to be small, as discussed in section 6.8.1.

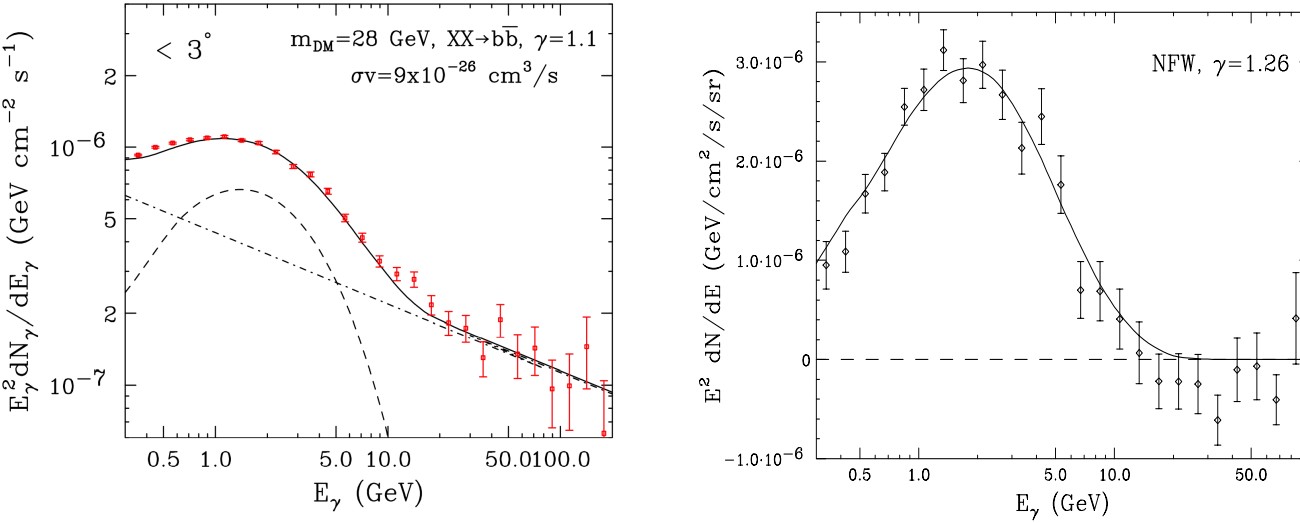

Figure 8.5: *The earliest (left) and the more recent (right) fits to the **GeV $\gamma$ excess in the Galactic Center** for DM annihilation to $b\bar{b}$. From Goodenough & Hooper (2009) and Daylan et al. (2014) in [745].*

Other alternative interpretations include the possibility of a spectral break in the emission of the central Black Hole and the possibility that past isolated injections of charged particles (electrons or protons, in one or more bursts, possibly connected with the activity of the central Black Hole), can produce sufficient amounts of secondary radiation to account for the anomalous signal. While reproducing all the details of the observed excess might be not easy with these models, they represent additional plausible counterexamples to the DM interpretation.

Testing the DM interpretation by searching for associated indirect detection signals is a challenging task. After various uncertainties are properly taken into account, neither the $\bar{p}$ constraints, the CMB observations, nor the $\gamma$-ray constraints from Fermi observations of dwarf galaxies can conclusively confirm or refute the DM interpretation of the GC GeV excess. Similarly, testing the DM interpretation through direct detection or collider searches is not straightforward. This may seem surprising, given that the properties of the best fit candidate ($\sim 45\,\text{GeV}$ DM coupling to quarks with weak-scale strength) naïvely guarantee a signal at colliders and in nuclear recoil experiments. However, reality is more complicated,[4] and many 'loopholes' are possible. For instance, indirect detection could proceed via a resonance in the $s$-channel, which would enhance the $\gamma$-ray signal but not the signals in direct detection and at the LHC. Indirect detection could also proceed via intermediate vector states, DM DM $\rightarrow VV \rightarrow b\bar{b}b\bar{b}$, where $V$ has a very small mixing with the SM photon. The $V$ decays into quarks, even if suppressed, will eventually occur on galactic scales and produce the $\gamma$-ray signal, while the direct detection scattering process, featuring the $V/\gamma$ mixing in the $t$-channel, would be extremely suppressed. More systematically, several studies [756], working either in model-independent effective frameworks or in specific models, have shown explicit examples in which Dark Matter can be 'coy', meaning that it can have a single detectable signature (in this case the annihilation into $\gamma$-rays) while escaping all the other searches.

**Excesses from Andromeda?** Given the interest garnered by the GC GeV excess, it is logical to speculate whether a similar signal might be detectable in our nearest galaxy M31 Andromeda [757]. Intriguingly, the Fermi satellite has detected gamma-ray emissions from Andromeda, where a 'central, extended and symmetric' emission was found. However, its intensity is a factor of 5 higher than what

---

[4]See the discussion in section 7.3 for a more complete and general assessment.

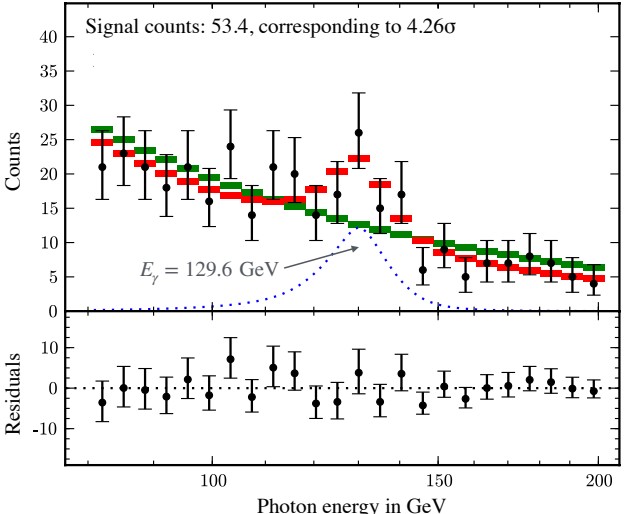 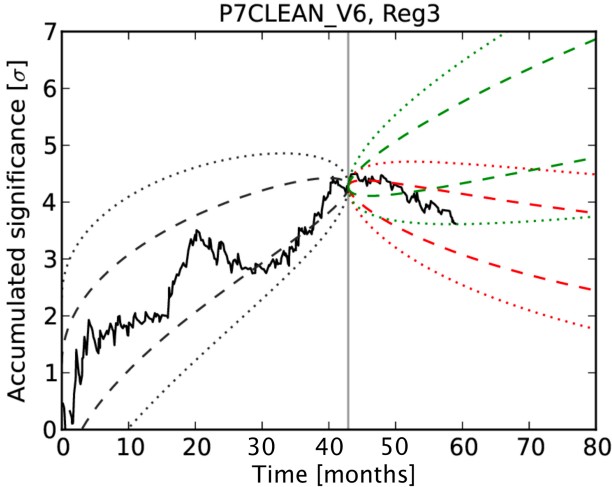

Figure 8.6: *Left:* Fermi *γ-ray data and fits pointing to* **a line at about 130 GeV**. *Right: behavior with time of the accumulated significance for this signal. The green lines correspond to the expected behavior of a real signal, with the dashed and dotted lines respectively associated with the 68%CL and 95%CL bands. The red lines trace the expected behavior of a statistical fluctuation. The black lines give extrapolations into the past for a constant source. Left figure recreated from data in* Weniger (2012); *right figure from* Weniger (presentation at the Fermi *meeting, 2013)* [742].

would be expected for a signal consistent with the MW GC excess. Another study (Karwin et al. (2019, 2020)) found evidence for an excess in the outer halo of Andromeda, roughly compatible in intensity and spectral shape with the MW GC signal. However, fitting it with DM requires a substructure boost factor of several orders of magnitude; the same boost factor applied to the MW would predict overwhelmingly large emissions from the MW halo, in contradiction with observations. A few recent studies, on the other hand, find no Andromeda excess attributable to DM, and thereby only derive upper limits.

### 8.2.6   A γ-ray excess from the Reticulum II dwarf galaxy?

In a period around 2015 it was claimed that several newly discovered dwarf galaxies (Reticulum II, and possibly Indus II, Tucana III and Tucana IV) could be showing a γ-ray excess around $2-10$ GeV, with up to $\approx 3\sigma$ statistical significance [737]. This was interpreted as annihilations of DM particles with estimates for DM mass ranging from a few GeV to a few hundred GeV, depending on the annihilation channel. The implied annihilation cross section was compatible with, or slightly higher than, the one required by DM as a thermal relic. These characteristics also made the putative signal compatible with the γ-ray GeV GC excess, discussed in the previous section. The analysis by Di Mauro et al. (2022) confirms the excess with similar significance, although other prior works had not [737].

### 8.2.7   A γ-ray line at 135 GeV?

Around the years 2012-2013 a claim about a '135 GeV line' in the photon energy spectrum gathered a lot of attention in the community [742]. It was originally identified by C. Weniger and collaborators in the publicly available Fermi data from an extended region that included the GC (the left panel in fig. 8.6 displays the most suggestive figure from the original analysis). The claim was later supported by the results of other analyses, with varying degrees of accuracy and claimed significance. The Fermi

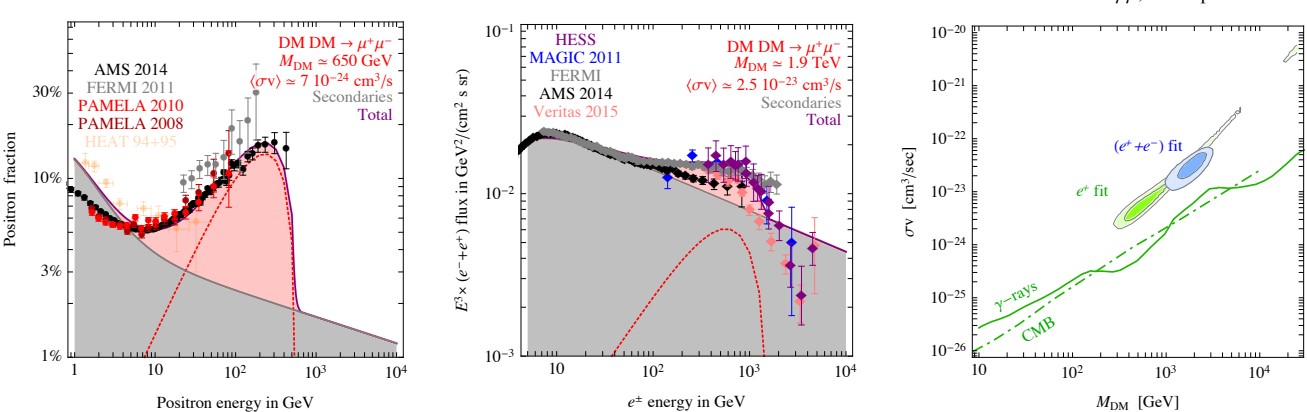

Figure 8.7:    *The* PAMELA/AMS ***positron excess*** *and its DM interpretation, circa 2015. Left: The annihilating DM fit of the rising positron fraction, featuring a leptophilic DM with a mass slightly below 1 TeV and a very large annihilation cross section. Center: The DM fit of the $e^+ + e^-$ flux, featuring a slightly heavier DM and an even larger annihilation cross section. Right: The regions in the DM parameter space preferred by the fit to only the positron fraction (green), and a fit to the $(e^+ + e^-)$ flux (blue), with contours denoting the $3\sigma$ and $5\sigma$ regions. The constraints from $\gamma$-rays (solid line) and the CMB (dot dashed) are those discussed in section 6.13.2.*

collaboration (Ackermann et al. 2013), for instance, claimed a $3.3\sigma$ ($1.5\sigma$) local (global) statistical significance for the existence of the line. Some studies observed the 135 GeV line in what could possibly be DM subhaloes of the MW and in galaxy clusters. Others found evidence of two lines, at 111 GeV and 129 GeV. Still others challenged the analyses, proposing that the line(s) could have an unidentified instrumental, statistical or astrophysical origin.

Tentative interpretations in terms of DM annihilation, and the associated DM model building, faced a significant challenge: if the line(s) were due to DM, it was plausible to expect an associated $\gamma$-ray continuum at lower energies, due to DM annihilations into other SM particles, as well as a signal in other CRs, such as anti-protons. The annihilation cross section into $\gamma\gamma$ required by the FERMI data, $\sigma v \approx \mathcal{O}(10^{-27})\,\mathrm{cm}^3/\mathrm{s}$, would generically imply a large $\gamma$-ray continuum flux in conflict with existing bounds. This argument did not, however, exclude all possible DM models, as shown by explicit examples.

With more accumulated data the signal significance diminished (see the right panel in fig. 8.6), thereby indicating that this was probably not an actual signal. An understanding gradually developed that the line had been a combination of an instrumental effect and a statistical fluctuation. The 2015 FERMI analysis of an increased and improved dataset [585] put the claim to its final rest as the local significance dropped to $0.7\sigma$. Additionally, the 2016 HESS search [584] also independently excluded it (albeit barely).

### 8.2.8   The PAMELA/AMS **positron excess**

In 2008, data from the PAMELA satellite [617, 618] revealed a steepening of the energy spectrum of the cosmic ray positron fraction, $e^+/(e^+ + e^-)$, above 10 GeV and up to 100 GeV (below $\sim 10$ GeV the situation is more involved, since the Sun distorts and modulates the inter-stellar spectrum, see page 209). This behaviour had previously been hinted at by the HEAT balloon experiment [616] and later confirmed by the FERMI satellite [619] and the AMS detector [620, 622]. AMS in particular extended the data to several hundreds GeV,[5] identifying a possible flattening above 300 GeV. The data, shown in fig. 8.7 (left),

---

[5]Published data on the positron fraction go up to 500 GeV, while for the $e^+$ flux they reach 1 TeV.

has the typical shape one would expect from DM, see the discussion in section 6, in particular fig. 6.1.

Given that the positron fraction in CR data grows with energy to almost 20%, interesting information may be provided also by the experiments that cannot discriminate $e^+$ from $e^-$, but are able to measure the energy spectrum of the electron–positron sum, $(e^+ + e^-)$, to higher energies. For this, several experiments performed around the same time showed hints of an anomalous 'bump' at a few TeV, possibly due to the same physics that generated the $e^+/(e^+ + e^-)$ excess, see fig. 8.7 (center). Both the FERMI satellite [625] and AMS [627] obtained a fairly featureless spectrum up to 1 TeV.[6] Above 1 TeV, data from the ground based Imaging Čerenkov telescopes HESS [624], MAGIC [626] and VERITAS [629] indicated a potential cutoff or a steepening of the spectrum at energies of a few TeV. Subsequently, the CALET and DAMPE collaborations also presented data: CALET in agreement with AMS, while DAMPE obtained a higher flux and steeper spectrum, see section 8.2.11 for further discussion.

In contrast to these intriguing excesses in the leptonic sector, data from PAMELA [643] (and, much later, AMS [645]) exhibited no anomalies at comparable energies in antiprotons. A compilation of the relevant data is shown in fig. 6.12.

The $e^+$ and $(e^+ + e^-)$ results deviate rather spectacularly from the power-law behavior expected from the cosmic ray propagation models. Importantly, they imply the existence of a source of *'primary'* $e^+$ (and $e^-$) beyond the already expected astrophysical sources.[7]

## DM interpretations

Quite naturally, DM annihilations (or decays) were immediately proposed as such a source [166]. It was quickly realized that in order for DM to fit the data, the DM particle had to:

▷ Have a mass in the TeV to multi-TeV range, to reproduce the feature in the $(e^+ + e^-)$ spectrum.

▷ Have an annihilation cross section of the order of $\sigma v \approx 10^{-23}\,\mathrm{cm}^3/\mathrm{s}$ or larger, in order to produce large enough $e^\pm$ flux to fit the positron rise and the $(e^+ + e^-)$ bump. This cross section is significantly higher than the annihilation cross section required for a thermal relic, $\sigma v \approx 2\,10^{-26}\,\mathrm{cm}^3/\mathrm{sec}$, see eq. (4.13).

▷ Be leptophilic, i.e., predominantly annihilate (or decay) into leptons to avoid contradicting the absence of an anomaly in antiprotons.

Fig. 8.7 illustrates the typical fits that lead to these conclusions.

The above requirements were a challenge for the 'traditional' DM candidates. Taking as a strawman example the supersymmetric neutralino (section 10.1.2), we can see that it scores poorly on all of these criteria. *i)* The required multi-TeV DM mass is in tension with the low scale superymmetry being the solution to the hierarchy problem, since this implies weak scale masses for supersymmetric partners; *ii)* The required large annihilation cross section is generically incompatible with the thermal DM production mechanism. Since the neutralino is a Majorana particle the *s*-wave annihilation cross section for annihilations into the $f\bar{f}$ final state are helicity suppressed by $(m_f/M)^2$, where $m_f$ is the fermion mass, giving a large suppression for light leptons. *iii)* The neutralino is not naturally leptophilic. The typically dominant neutralino annihilation channels, such as the annihilations to gauge bosons, do not distinguish between leptons and quarks, and thus lead to too large amounts of hadrons.

---

[6]After initial mild disagreement in the spectral indexes (AMS being steeper), the FERMI collaboration released new Pass 8 data in 2017 [630], which brought its data points closer to those of AMS.

[7]Simple kinematic arguments predict that the spectrum of any *secondary* positrons should decrease much more steeply than observed, see Serpico (2012) [758]. In this context, *primary* refers to the charged CRs produced directly by some (astrophysical or exotic) source while *secondary* refers to the CRs generated as byproducts of primaries colliding with the gas particles of the interstellar medium.

These generic arguments could be circumvented in specific situations, but at the price of making at least some fine-tuned choices: about the $\bar{p}$ background properties, requiring a nearby DM clump, assuming (uplifted) resonances, finding just a few configurations in the scan of the DM parameters space, or explaining only the positron rise and leaving the rest to ad-hoc astrophysics. Similar arguments applied to other 'traditional' DM frameworks, e.g., the Kaluza-Klein (extra-dimensional) DM (section 10.1.3).

As the result, the community shifted towards exploring new model-building possibilities. A recurrent theme was how to reconcile the large annihilation rate 'today' with the smaller rate required by the thermal DM production in the early universe. Such a possibility of a large flux is of course interesting more broadly, since it greatly increases the hope for a detection. Historically, the PAMELA/AMS excesses marked the moment when contemplating large DM annihilation signals became socially acceptable. The main strategies for achieving such an enhancement are:

a) Via an **astrophysical boost factor** due to DM overdensities in the galactic halo (see section 6.8.1). Typical realistic values, however, have been demonstrated to reach at most $\mathcal{O}(30)$.

b) Through **annihilation via a resonance**. If the resonance mass is just below twice the DM mass, the annihilation cross section becomes sensitive to the details of the DM velocity distribution. Since DM particles are on average slower today than in the Early Universe, many more satisfy the conditions for resonantly enhanced annihilation, provided that the width of the resonance is also appropriately fine tuned.

c) Leveraging the **Sommerfeld enhancement** (see section 6.8.2): the enhancement is inversely proportional to the relative velocity of annihilating DM particles, and is thus larger today than in the Early Universe.

The intense activity aimed at explaining the CR excesses falls into several categories [759]:

◇ **Minimalistic Dark Matter models** introduced the fewest new particles beyond the Standard Model while still providing a viable DM candidate. Notable examples include Minimal Dark Matter (MDM, section 9.3.4), the Inert Higgs (IDM, section 9.3.5), among others. The MDM model contains in its minimal realization just the SM and a multi-TeV DM particle, annihilating almost exclusively into $W^+W^-$. Initially, its authors were very excited since it had predicted the correct size and shape of the positron rise in PAMELA, as well as the null result in $\bar{p}$, provided that an astrophysical boost factor of $\sim 50$ was adopted. Later the MDM model was disfavored by the $e^{\pm}$ data, which prefer a lower mass, and by the $\gamma$-ray data, see e.g., Pato et al. (2009) in [760]. The phenomenology of the IDM shares a similar history, see Nezri et al. (2009) in [759].

◇ **Models with new dark forces** or, more generically, a rich dark sector (see section 9.4). Most of the models whose construction has been directly stimulated by the CR excesses fall into this class. The model that has undeniably garnered the most attention was introduced by Arkani-Hamed et al. (2008) [759], although similar ideas have been proposed either earlier or at around the same time. The model features a TeV-ish DM particle, not charged under the SM gauge group, but with self-interactions mediated by a new force-carrying boson $\phi$ (with the interaction strength comparable to typical gauge couplings). A small mixing between $\phi$ and the electromagnetic current ensures that $\phi$ eventually decays. DM annihilations proceed in two steps, the first step is the DM DM $\rightarrow \phi\phi$ annihilation, which is then followed by $\phi$ decaying into the SM particles. A crucial ingredient is that $\phi$ is chosen to be light, with the mass below $\lesssim 1$ GeV. This assumption ingeniously serves a dual purpose. Firstly, the $\phi$ exchanges induce the Sommerfeld enhancement. Secondly, due to its light mass, $\phi$ can only decay into SM particles lighter than a GeV, i.e., to electrons, muons and possibly pions, but not protons. This ensures that the annihilation is leptophilic, due to a simple kinematical reason. Other possibilities discussed in the literature for ensuring a leptophilic DM include symmetry based arguments, e.g., that DM carries a lepton number.

◇ **Decaying dark matter.** The possibility that the DM particles could decay on very long timescales has been entertained for quite some time, such as within the context of gravitino DM with $R$-parity violation (see section 10.1.2). Following the observations of the CR excesses, this idea has garnered renewed interest. The data can be fit, if the decay lifetime is close to $\approx 10^{26}$ seconds, and if the production of hadrons is adequately suppressed by some a priori unrelated mechanism.

## Problems with DM interpretations

Over time, the interest in interpreting the anomalies as effects of DM faded away, for a number of reasons. First of all, there are tensions in the data even when restricting to just the leptonic data: in fig. 8.7 the green and blue regions overlap only marginally, i.e., the positron data tend to prefer a smaller DM mass than the $(e^+ + e^-)$ data. This is due to the changing slope in the Ams-02 data at the highest energies. Furthermore, the updated data from Fermi [630] and Hess [631], along with additional observations from Dampe and Calet [732], have cast doubt on the distinct 'cutoff' at a few TeV, eroding the support for the coherent DM picture outlined above.

Secondly, DM interpretations faced increasing tensions with the constraints from gamma rays (discussed in section 6.13.2) and from the CMB (discussed in section 6.11.2) [760]. The constraints from gamma-ray emissions in the galactic halo were the first ones to be considered, including the secondary emissions from the inverse Compton process. For a peaked DM galactic profile, such as the commonly assumed NFW profile, these were found to exclude the interpretation of the leptonic excesses as annihilating DM, while for a cored DM profile the DM interpretation was still possible. The DM interpretation was subsequently excluded by the gamma-ray constraints from dwarf galaxies and from the extragalactic environment, as reported in fig. 8.7 (right). These are still the most stringent constraints (see section 6.3), but one should keep in mind that they are affected by the uncertainties regarding the content and the distribution of DM in the dwarfs (see section 2.2.3). The CMB constraints were found to exclude the DM interpretation in the case of $s$-wave annihilation, but not in the case of $p$-wave annihilation. Decaying DM is more solidly excluded, notably by the extragalactic $\gamma$-ray background measurements and the galactic halo constraints.

The final conclusion is that it is technically still possible to fit the leptonic excesses using annihilating DM, e.g., in the $\mu^+\mu^-$ channel, and even slightly better in the 4 lepton channels (these arise in secluded DM models). However, there are a number of stringent conditions that need to be satisfied: on top of the global fit being rather poor, one needs to invoke significant uncertainties in order to relax the dwarf gamma-ray limits, assume a cored DM profile to circumvent the galactic gamma-ray bounds, assume $p$-wave annihilation to evade the CMB constraints,... Since the electrons and positrons do not propagate very far outside their galactic neighbourhood, it is also possible that the solar system is in a region of the Milky Way with higher DM density. For instance, we could be inside a DM sub-halo, which would then enhance the signal but be irrelevant for most constraints.

## Astrophysical interpretations

Last but not least, there are plausible astrophysical interpretations of the $e^\pm$ excess that, at the moment, appear less contrived than the explanations invoking DM [758]. A hypothesis widely discussed from the very beginning when excesses were observed, and already well-known within the astrophysical community even before that, centers on **pulsars** (rapidly spinning neutron stars). The very intense magnetic field present in these objects can strip electrons from the neutron star's surface. These electrons then emit high-energy $\gamma$'s, which efficiently produce $e^\pm$ pairs. Trapped in the surrounding cloud, these particles undergo further acceleration before their eventual release into the interstellar medium. Both the energy budget and the spectral shape are well suited to explain the data, with the mechanism inherently producing few hadrons, in line with the observed 'leptophilic' nature of the excesses. The pulsar responsible for the

excess should be relatively 'nearby', less than a kpc, and 'young', less than a few hundred kyr. A more distant or older star would result in excessive $e^\pm$ diffusion in the galactic medium, leading to a flux that is both too small in magnitude and of too low-energy. Nearby pulsars can generate an $e^\pm$ excess around our location without affecting the entire Milky Way, thus avoiding the bounds from the secondary gamma rays. Detailed fits found that it is unlikely that the excess can be explained fully by a single pulsar, while a combination of a few local and/or numerous galactic pulsars could provide a very good explanation of the $e^\pm$ data.

Other proposed astrophysical explanations include the production and acceleration of positrons in old, nearby **Supernova Remnants** (SNR). The origin of these positrons would be the decays of charged pions, which are created from interactions of cosmic ray protons accelerated within the SNR environment. Since the interactions are hadronic, the simplest realization of this mechanism also predicts a so far unconfirmed hardening feature in the antiproton cosmic ray flux at high energies.

Some studies have also challenged the primary nature of the positron rise and proposed that it is instead made only of **secondaries**, i.e., the $e^+$ produced by the collisions of high energy primary CRs with the ambient interstellar matter. The 2009 model by Katz, Blum and Waxman had a clean prediction for the positron fraction to saturate at $\approx 20\%$ at the energy of $\approx 300$ GeV, in agreement with the subsequent 2013 Ams data. On the other hand, these models typically assume a thin diffusive halo and negligible radiative losses for $e^\pm$, which is at odds with what is commonly assumed.

In the case of a single source such as a pulsar or a SNR, or in the case of a few clustered sources such as the astrophysical sources in the galactic disk, the positron flux would be slightly directional and may result in a statistically significant **anisotropy**. Its lowest mode, the dipole, is expected to range from a few ‰ to about 10% depending on the energy (energetic $e^\pm$ are less subject to direction-randomizing diffusion). In contrast, DM is a distributed source that is predicted to produce an essentially isotropic flux: the dipole induced by the concentration towards the GC is below the few ‰ level at the relevant energies. Anisotropy measurements can therefore discriminate between the origins of the positrons. Current measurements from Fermi [761] and Ams [641] though only impose upper bounds on the dipole, at the level of a few percent (1.9% at energies above 16 GeV for Ams).

DM interpretation made a short come-back in 2017 [762]. The Hawc telescope detected an extended $\gamma$-ray emission at TeV energies (previously also detected by Milagro) around Geminga and Monogem, two local pulsars routinely considered to be plausible emitters of the excess positrons. Interpreted as a high-energy inverse Compton scattering emission, this would indicate that the pulsars indeed produce highly energetic $e^\pm$. The question remains, though, whether or not these $e^\pm$ remain trapped in the pulsars' surroundings for longer periods of time and cannot reach the Earth with large enough energies. If the claims of efficient confinement are correct then pulsars cannot account for the excesses, reintroducing DM as a potential source. Subsequent analyses using a more refined description of $e^\pm$ diffusion in the local galactic environment, however, support the possibility of pulsars being the source of the excess positrons.

The overall conclusion from the above investigations is that the $e^+$ flux, being contaminated by the local astrophysical backgrounds, may not be as effective a probe for detecting DM as it was initially anticipated.

### 8.2.9 An anti-proton Ams excess?

Several studies have reported an excess of antiprotons relative to the predicted astrophysical background, interpreting it as a result of DM annihilations [739]. The initial claim was based on Pamela data, while the more recent claim is based on the Ams data released in 2015. The putative excess is at kinetic energies of approximately 10 to 20 GeV and can be fit by an 80 GeV DM particle annihilating into $b\bar{b}$ with a roughly thermal cross section (see the left panel in fig. 8.8 for an illustration[8]). The fact that

---

[8]We acknowledge the help of M. Boudaud for the figure.

these properties are typical for vanilla WIMP DM and are so close to the ones needed to explain the GC GeV excess in gamma-rays (section 8.2.5), makes these findings even more interesting.

On the other hand, the predictions of low energy antiproton cosmic ray spectra are affected by significant uncertainties, mostly related to which propagation model is assumed and how its parameters are determined, how the solar modulation effects are modeled, which antiproton production cross sections are assumed, and how the correlations among data are handled. Several comprehensive analyses (Reinert et al. (2018), Boudaud et al. (2019), Heisig et al. (2020), Calore et al. (2022), De La Torre Luque et al. (2024) in [739]) concluded that the data are consistent with a purely secondary origin and that the global significance of the excess drops to less than $\sim 2\sigma$ once all the uncertainties are taken into account. Further reduction in these uncertainties will thus provide more insight into any possible excess.

### 8.2.10   An anti-helium AMS excess?

Beginning in late 2016, the AMS collaboration has announced, through press releases and presentation slides (though without published papers), their observation of several events compatible with anti-helium [430]. As of 2023, reports indicate 9 events, with no clear distinction between $^3\overline{\mathrm{He}}$ and $^4\overline{\mathrm{He}}$. Previous reports citing 8 events pointed to 6 and 2 candidates, respectively. These observations occurred over approximately 10 years of data collection.[9] While no explicit information about the energy is given, one can infer from the rigidity measurements that the events indicate rather large values ($K/n \simeq 6$ GeV/$n$ to 20 GeV/$n$).

If confirmed, this would point to a surprising and unexpected production of anti-helium, either from astrophysics or from DM. While DM annihilations or decays can in principle produce anti-helium nuclei via coalescence of $\bar{p}$ and $\bar{n}$ (see sections 6.6 and 6.13.2), the rate is highly suppressed as it requires 3 (4) nucleons to coalesce for the production of a $^3\overline{\mathrm{He}}$ nucleus ($^4\overline{\mathrm{He}}$). Initial calculations [430] indeed predicted a DM induced anti-helium flux that is many orders of magnitude below the estimated sensitivity of AMS, especially after the constraints from not producing too many antiprotons are taken into account. Similarly, the flux of anti-helium produced in astrophysical process is also expected to be very small, though closer to the reach of the experiments. These predictions have large uncertainties, encoded in the value of the coalescence momentum $p_0$, which enters the predictions with a high power. Several works [430] have thus argued that large enough DM or astrophysical production of anti-helium is possible in principle, e.g., by increasing the coalescence parameter to values much larger than initially thought or by including some previously neglected standard model processes.

The existing upper bounds on the anti-helium fluxes come from AMS-01 [660], PAMELA [661] and BESS-POLAR II [662], see section 6.13.2.

### 8.2.11   An electron+positron DAMPE excess?

In 2017 the CALET [516] and DAMPE [518] experiments released their measurements of the 'all leptons' spectrum ($e^+ + e^-$) [632,633], the zoomed in version of which is shown in fig. 8.8 (right). The CALET data are in good agreement with the previous measurements of the same quantity by AMS, while the DAMPE data indicate a higher flux and a harder spectrum. The source of this discrepancy remains unclear, and could be due to an incorrect energy calibration of one, two or all three of the experiments (all three have to rely on extrapolations above $E \approx 100$ GeV, since for such high energies the test beam data are not available).

While the CALET data have remained relatively unexploited, various papers interpreted the DAMPE data in terms of DM [732]. Two main features attracted attention: *a)* the hint of a spectral break around 0.9 TeV; *b)* the single DAMPE data point at $E \simeq 1.4$ TeV, which formally sits 2.6 $\sigma$ (global) above the

---

[9]There are also reports of 7 anti-deuteron events, see P. Zuccon (2022) [430].

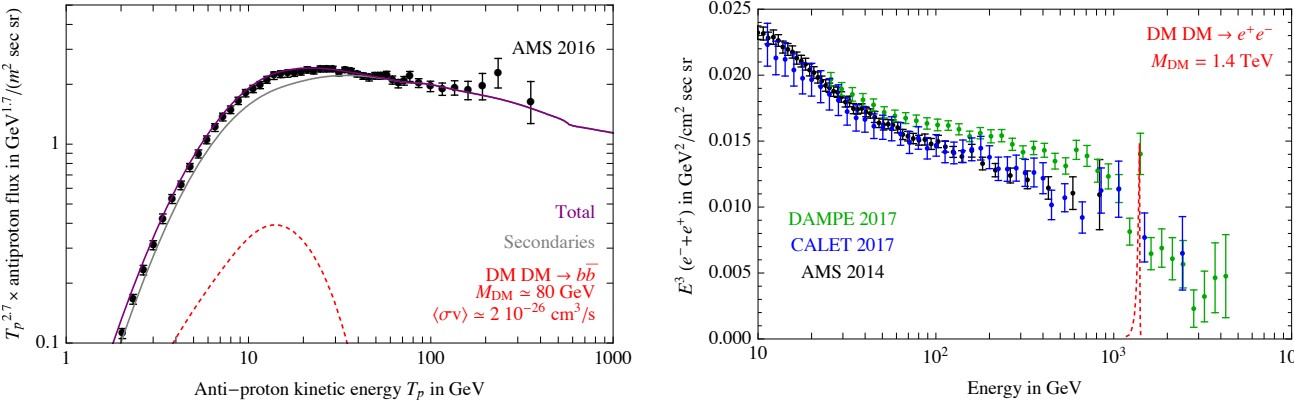

Figure 8.8: *Illustration of* **recent claimed excesses in charged cosmic ray data**. *Left: potential antiproton excess in* Ams *data. The grey line represents the astrophysical flux of secondary p̄ that, together with a DM contribution (red dashed line), would best fits the data according to recent studies. The secondary p̄ best fit flux without DM is not shown. Right: possible lepton excess in* Dampe *data.*

power law spectrum prediction. The point a) is not qualitatively different from the knee-like feature discussed in section 8.2.8. Fitting b) in terms of DM requires DM annihilating or decaying into $e^+e^-$, inside a very large ($\sim 10^8\ M_\odot$) and very close clump (within 0.3 kpc). Any other channel and/or a more distant clump (or a smooth DM distribution) would not result in such a sharp spectrum.

## 8.2.12 The ANITA anomaly

ANITA, a balloon-borne radio antenna, has conducted several one-month-long flights over Antarctica since 2006 [507]. It was primarily designed to detect ultra-high-energy neutrinos via the Askaryan radiation which these emit when hitting the ice (the effect can be thought of as a collective Čerenkov radiation from the particles in the compact shower initiated by the neutrino). ANITA was also sensitive to the radio emission of Extensive Air Showers (EASs) produced by ultra-high-energy particles in the atmosphere: either ordinary cosmic rays from the upper atmosphere, or $\tau$ leptons decaying in the air after production in the ice layer by a $\nu_\tau$ that has skimmed the Earth (the focus is on the $\tau$ flavor since electrons and muons are absorbed before emerging from the ice).

ANITA reported the detection of two puzzling events which appear to be *upward*-traveling and very energetic EASs ($\approx$0.6 EeV in both cases) [731]. They are puzzling because $\tau$ neutrinos with such extreme energies interact too much with matter and thus cannot traverse the Earth, even taking into account $\tau$-regeneration (see section 6.9.2).[10] Since dark sector constructions can include particles that interact with matter less than neutrinos do, various DM-related interpretations have been put forward. In these models, super-heavy DM particles (e.g., with $M > 10^{10}$ GeV) decay into lighter, extremely boosted particles that are then responsible for the observed signal. Other new-physics interpretations, unrelated to DM, have also been proposed, based on hypothetical particles that are less interacting than neutrinos, but possibly produced through neutrino conversion.

In 2020 ANITA reported a detection of four additional very high-energy ($\approx$ EeV) events, which are not upward-traveling but rather horizon-skimming [763]. These are not puzzling by themselves, since they can be produced by (unusual but not impossible) very-high-energy $\tau$ neutrinos. However, such a SM

---

[10]An alternative and more mundane interpretation posits that the signal is the reflection on the ice surface of 'ordinary' downward cosmic ray events, not correctly reconstructed by the experiment.

explanation is in tension with the limits imposed by other observatories such as Auger and Icecube. These events have thus also been interpreted in terms of decaying DM with an EeV mass.

### 8.2.13   A Gamma-Ray Burst 221009 anomaly?

In 2022 two experiments claimed the detection of unusually high-energy photons about 1 hour after the GRB 221009A $\gamma$-ray burst happened at high red-shift $z \approx 0.15$. Lhaaso, in China, claimed an observation of photons up to 18 TeV (later revised to 13 TeV); Carpet-2, situated at Baksan in Russia, claimed an air shower consistent with a 251 TeV photon [723]. If the temporal coincidence means that the photons come from the extragalactic $\gamma$-ray burst rather than from a nearby source, this poses a puzzle: the photon energies are above the threshold for absorption on the extragalactic background light, as discussed in section 6.2.3. That is, the optical depths for photons originating from $z \approx 0.15$ are $\tau_{\rm opt} \approx 10^4$ at $251\,{\rm TeV}$ and $\tau_{\rm opt} \approx 14$ at $18\,{\rm TeV}$ (although the uncertain star-light contribution is dominant at this lower energy). The resulting $e^{-\tau_{\rm opt}}$ suppression of the flux is certainly too large for the 251 TeV photons, and possibly also for the 18 TeV photons.

According to some authors [723], the absorption can be avoided or mitigated in the presence of new physics, such as photon/axion oscillations where the axion has a mass $m_a \sim 10^{-10}\,{\rm GeV}$ and the coupling to photons is $g_{a\gamma\gamma} \sim 10^{-11}/\,{\rm GeV}$ (axion as a DM candidate is discussed in section 10.4).

### 8.2.14   A 220 PeV neutrino?

In 2025, the Km3NeT collaboration announced the detection of a cosmic-ray neutrino event of apparent extragalactic origin, with an estimated energy of approximately $220\,{\rm PeV}$ [720]. This energy is an order of magnitude higher than that of neutrinos observed by the similar IceCube experiment The event might just be a $\sim 3\sigma$ fluctuation in the observation of neutrinos of astrophysical origin. However, some authors interpret it as evidence of a spectral hardening in the cosmic-ray neutrino energy distribution, potentially due to an additional component arising from the decay of super-heavy DM. [720]. The required DM mass would be in the range of hundreds of PeV, with a lifetime around $10^{29}\,{\rm sec}$, close to the indirect detection bounds for typical decay channels involving neutrinos (see fig. 6.15).

## 8.3   Collider/accelerator: current anomalies

Several anomalies seen in data from colliders and accelerators admit possible new physics interpretations that could include DM. Sometimes the connection to DM is direct, for instance the neutron decay anomaly (section 8.3.3) can be explained by requiring a sizable decay rate to invisible states. In other cases, for instance in the $(g-2)_\mu$ and the $R_{K^*}$ anomalies (sections 8.3.1 and 8.3.2 respectively), the connection with DM is indirect, with DM possibly contributing through one-loop corrections.

### 8.3.1   The $(g-2)_\mu$ anomaly

The anomalous magnetic moment of the muon, $a_\mu = (g-2)_\mu/2$, is a precisely measured quantity with experimental and theoretical errors below the $10^{-6}$ level. The measurement by the Fermilab Muon $g-2$ Experiment [764] confirmed, with improved precision, the 2006 measurement by the BNL E821 experiment at Brookhaven [746]. QCD loop corrections to the photon propagator induce the dominant uncertainty on the theoretical predictions of $a_\mu$. These effects have been computed in two ways:

1. Via dispersion relations using $e^+e^-$ scattering data, with the dominant contribution from $e^+e^- \to \pi\pi$ close to threshold. This technique was adopted for the so called *consensus* SM prediction [765], which shows a $5\sigma$ tension with the experimental results. The issue is not settled, however, with more recent $e^+e^-$ scattering data from CMD-3 implying no anomaly [766].

2. Via lattice QCD computations. This technique shows agreement with the experiment [767].

A deviation in $\Delta a_\mu$ could have implications for DM models. Both the SM and new physics contributions to $a_\mu$ start at 1-loop. The new physics interpretations can thus include DM running in the loop, if DM has couplings to muons. The models for new physics explanations of $\Delta a_\mu$ fall into two broad categories. In the first category the chirality flip required by $\Delta a_\mu$ is due to the muon mass insertion on the external muon leg. The new physics contribution to $a_\mu$ is thus heavily suppressed by the muon mass, so that the new physics that explains the deviation needs to be light. The most prominent example is a light $\mathcal{O}(100\text{ MeV})$ gauge boson, for instance due to a new $L_\mu - L_\tau$ gauge group.

In the other class of models the chirality flip arises from the Higgs coupling to the internal states running in the loop. New physics can have a large coupling to the Higgs, and can thereby be heavy, up to a few TeV, and still explain the $a_\mu$ anomaly. If the new physics states running in the loop are odd under some $\mathbb{Z}_2$ symmetry, the lightest state is stable, and can be a DM candidate. In fact there are many such DM models that even lead to the right DM thermal relic abundance [768].

## 8.3.2 The flavour anomalies

In the period 2014-2022, some deviations from the SM have been claimed in rare processes corresponding to the quark level transition $b \to s\mu^+\mu^-$. The $b$ and $s$ quarks are bound inside hadrons, so not all observables are theoretically clean, and may involve non-perturbative QCD effects. However, the muon-to-electron ratios such as $R_K = \text{BR}\left(B^+ \to K^+\mu^+\mu^-\right)/\text{BR}\left(B^+ \to K^+e^+e^-\right)$ (and similar for $K_S$, $K^{*+}$) are theoretically clean. Away from the kinematical thresholds such ratios are predicted in the SM to be very close to 1, since the electroweak interactions do not distinguish between different lepton generations. The only symmetry-breaking effects are due to different muon and electron masses, and can be easily taken into account in theoretical predictions.

The most precise measurements of these quantities are due to the LHCb experiment at CERN, which originally claimed $R_{K^{(*)}}$ values around 0.8 [740]. Global fits to both $R_{K^{(*)}}$ as well as the $b \to s\mu^+\mu^-$ branching ratios and angular distributions in several decay channels then implied a statistical significance of the anomaly of about $5\sigma$ [769]. These anomalies could be fitted by adding to the SM the effective operators of the form $(\bar{s}\cdots b)(\bar{\ell}\cdots\ell)/(30\,\text{TeV})^2$, where the ellipses denote appropriate chiral structures. The operators could be mediated by new TeV states, including DM. In the 2022 analysis the LHCb collaboration improved their treatment of systematic effects in the measurement, which then resulted in revised $R_{K^{(*)}}$ values close to 1, in agreement with the SM. The remaining anomalies are now only in the less easily predictable observables affected by the non-perturbative QCD effects.

There is also a hint of anomalies in a different set of theoretically clean ratios, $R_{D^{(*)}} \equiv \text{BR}(\bar{B} \to D^{(*)}\tau\bar{\nu}_\tau)/\text{BR}(\bar{B} \to D^{(*)}\ell\bar{\nu}_\ell)$. However, their statistical significance is less notable.

## 8.3.3 The neutron decay anomaly

At low energies, DM has been suggested as a possible explanation of an anomaly in the neutron lifetime measurements. The neutron undergoes weak $\beta$ decays, where the dominant decay channel, $n \to pe\bar{\nu}_e$, already has a highly suppressed decay rate $\Gamma_n^\beta \sim G_F^2(m_n - m_p)^5$. The neutron lifetime is measured using two different experimental techniques, which consistently give results that differ at about $4\sigma$ level [729], as summarized in fig. 8.9:

- The **'bottle' method**. Ultra-cold neutrons are stored in a magnetic bottle via a small trapping potential, $V_{\text{trap}} \approx 50\,\text{neV}$. After some time $t$ the neutrons remaining in the magnetic bottle are counted, fitting the result to $N(t) = N_0 e^{-\Gamma_n^{\text{tot}}t}$. This measures the total neutron decay width, giving $\Gamma_n^{\text{tot}} = 1/(879.4 \pm 0.6)\,\text{s}$.

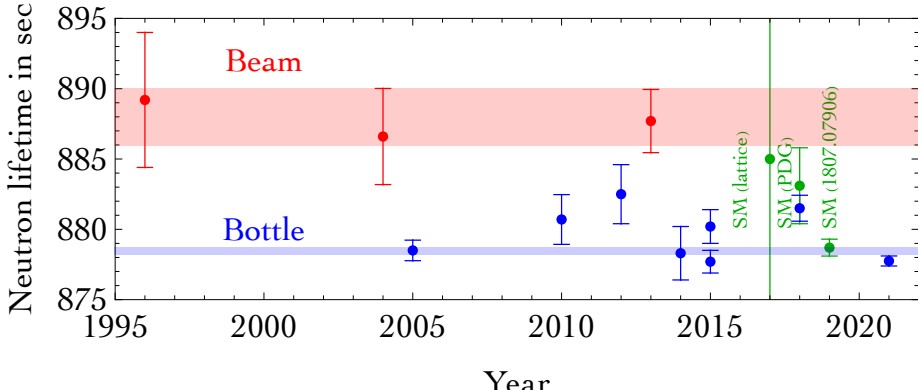

Figure 8.9: *'Bottle' (blue) and 'beam' (red) determinations of the **neutron lifetime**, compared to the SM prediction (green).*

- The **'beam' method**. A beam of cold neutrons is monitored by counting the protons from the neutron $\beta$ decays, and fitting the result to $dN/dt = -\Gamma_n^\beta N$. The measured decay rate, which is here entirely attributed to $\beta$ decays since it is signalled by the counted protons, is found to be smaller, $\Gamma_n^\beta = 1/(888 \pm 2)\,\text{s}$.

The SM predicts $\Gamma_n^{\text{tot}} = \Gamma_n^\beta$, up to known additional decay channels, which contribute at the 1% level. The neutron decay anomaly, i.e., the mismatch between $\Gamma_n^{\text{tot}}$ and $\Gamma_n^\beta$, could be due to an entirely new decay channel, where the neutron decays into nearly invisible particles, including DM. The most discussed possibility is $n \to \text{DM}\,\gamma$ with $M \lesssim m_n$; $n \to 3\text{DM}$ with $M \lesssim m_n/3$ is also possible [729]. Alternatively, for large enough cross sections, the neutrons stored in a bottle could be kicked out by soft scatterings with DM particles, which were captured and thermalized in the Earth [729].

The above interpretations are in tension with the precise SM prediction for the neutron decay rate, $\Gamma_n^{\text{SM}} = 1/(878.7 \pm 0.6)\,\text{s}$. This agrees with the experimental 'bottle method' result and thus, if correct, leaves no room for the type of new physics contributions that increase the decay rates, indicating a problem with the 'beam' measurements. The caveat is that the quoted precise SM prediction employs the determinations of $V_{ud}$ and the PERKEO III measurement of the neutron axial coupling constant $g_A$ [770], which are potentially problematic at the claimed level of precision. The less precise aSPECT [770] measurement of $g_A$, for instance, favours the value of $\Gamma_n^{\text{SM}}$ that is close to the beam experiment result. This additional layer of experimental uncertainty adds confusion to the subject [729].

An entirely different possibility, which does not run into issues with the $\Gamma_n^{\text{SM}}$ problem, is that the neutron decay anomaly is due to a neutron oscillating into a quasi-degenerate DM state [729]. A mass splitting between the normal neutron and the DM state of order $\Gamma_n^{\text{tot}} - \Gamma_n^\beta$ is needed. The DM state then decays invisibly with the same lifetime as the neutron (a possible reason for this would be that DM is a mirror neutron, $n'$, decaying via the $n' \to p'e'\bar{\nu}'_e$ channel [771]). The net result is a deficit of protons measured in the beam experiments, while there is no modification of the neutron lifetime when measured using the bottle method. The above interpretation is consistent with the experimental bounds on the loss of neutrons through the wall of the bottle, if the strong magnetic fields $B \approx \mathcal{O}(1)\,\text{T}$, used in the beam experiments, enhance oscillations via resonance effects.

All the new physics models need a large coupling between neutron and DM. This implies that DM is produced inside neutron stars. The resulting equation of state is softened and tends to be incompatible with the existence of heavy neutron stars, which are observed to have masses up to 2 solar masses.[11] Possible ways out are that the $n \to 3\,\text{DM}$ decay channel is open (this has a milder effect on the equation

---

[11]The physics reason behind it is the same as for the Oppenheimer and Volkoff's calculation in 1939, in which

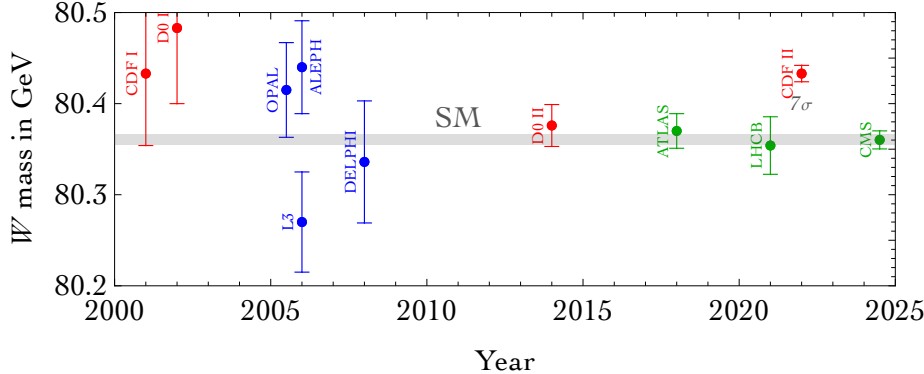

Figure 8.10: *W-mass measurements* from LEP (blue), LHC (green), and TeVatron (red), compared to the SM prediction (gray) from global electroweak fits.

of state, and thus remains compatible with data), or that DM exhibits a strong repulsive force with itself or with neutrons (this makes it expensive to produce DM particles in a purely baryonic medium, both from an energetic point of view and from a chemical potential one) [729]. In the latter case, the required DM scattering cross section is comparable with the typical QCD cross sections, and thus risks being in conflict with the bullet-cluster bound on DM interactions, eq. (1.5).

### 8.3.4 The $W$-mass anomaly

In 2022, the CDF TeVatron collaboration reported a measurement of the $W$-boson mass $M_W$ that is $0.7\,\%_{\rm oo}$ higher than the SM prediction [724]. The statistical significance of this tension is $7\sigma$. The CDF result was also in $\approx 4\sigma$ tension with the other less precise direct experimental determinations of $M_W$. In 2024 the CMS LHC collaboration reported a measurement of $M_W$ with precision comparable to CDF, but in agreement with all the other measurements, and also with the SM prediction. The most plausible interpretation is that there is a problem with the CDF result.

Alternatively, the CDM $M_W$ anomaly can be accommodated by extending the SM Lagrangian by a single dimension 6 operator $\mathscr{L} = \mathscr{L}_{\rm SM} - (H^\dagger D_\mu H)^2/(6\,{\rm TeV})^2$. The extra operator can be mediated at tree level by multi-TeV extra scalars that obtain vacuum expectation values or by an extra $Z'$ vector boson coupled to the Higgs, $H$. Alternatively, it can arise at one loop level in the presence of new ${\rm SU}(2)_L$ multiplets with weak scale mass, but with the masses of different multiplet components significantly split by interactions with the Higgs. This latter kind of new physics is in general significantly constrained by the LHC collider data. The collider constraints get relaxed if the decays of the multiplet components involve a neutral invisible particle, which can play the role of DM [724].

### 8.3.5 The $\gamma\gamma$ peak at 750 GeV and other LHC anomalies

In 2015, at the beginning of the $\sqrt{s} = 13\,{\rm TeV}$ Run2 at the LHC, the data taken by the ATLAS and CMS collaborations hinted at an excess in the $pp \to \gamma\gamma$ channel, peaked at an invariant mass of around 750 GeV, and with a local statistical significance of about $4\sigma$ [734]. This unexpected result sparked a plethora of theoretical speculations and interpretations. Given that the new particle would have had a weak-scale mass, it could have mediated interactions with DM, producing the desired DM thermal relic

---

they obtained an *underestimated* value for the maximal mass of neutron stars, because they neglected neutron–neutron interactions. The pressure is lowered at a given energy density because DM replaces neutrons and reduces their Fermi momentum and pressure.

abundance via the freeze-out mechanism. However, the excess did not reappear in the next round of data taking, and was soon dismissed as a statistical fluctuation.

At the LHC so many different distributions are measured that on a statistical basis one expects quite a few anomalies at the few $\sigma$ level. Over the course of its operation, numerous such anomalies have indeed been observed, only to subsequently disappear as the integrated luminosity doubled. The LHC data hint at:

⤳ a $\gamma\gamma$ resonance at 95 GeV at 152 GeV and at 680 GeV [772];

⤳ resonant $t\bar{t}$ production around 400 GeV [772];

⤳ di-jet production around 1 TeV and 3 TeV [772];

⤳ non-resonant excess of $e^-e^+$ [772];

⤳ some distributions of multi-lepton events which seem not to agree well with SM backgrounds [772];

⤳ events with soft leptons plus missing energy indicate tentative excess that could be interpreted as DM [773].

More data can clarify if these are all mere statistical fluctuations.

### 8.3.6   The ATOMKI excess at 17 MeV

In 2015, the ATOMKI experiment conducted measurements of the $e^\pm$ particles produced in the decay of excited $^8$Be$^*$, with $J^P = 1^+$, into the $0^+$ ground state. The ATOMKI team reported a $\approx 7\sigma$ evidence for an excess broadly peaked around $\theta \sim 135°$, where $\theta$ is the angle between the outgoing $e^-$ and $e^+$ (their energies and charges were not measured) [738]. The excess can be interpreted as an on-shell production of a new boson $X$ with mass $(17 \pm 0.3)$ MeV and

$$\mathrm{BR}(^8\mathrm{Be}^* \to X\,^8\mathrm{Be} \to e^-e^+\,^8\mathrm{Be}) \approx 0.6\ 10^{-5}\,\mathrm{BR}(^8\mathrm{Be}^* \to \gamma\,^8\mathrm{Be}). \tag{8.3}$$

Subsequently, in 2021 and 2022, the ATOMKI team identified a similar anomaly at roughly the same mass in the decays of $^4$He$^*$ and $^{12}$C$^*$, respectively [738]. In 2023 a consistent result was obtained by a combined team of ATOMKI and Vietnamese researchers (Ahn (2023) in [738]). Furthermore, the $\gamma\gamma$ resonances at 17 and 38 MeV were reported by Abraamyan et al. (2023) [738]. The MEG-II experiment at PSI did not see a signal, disfavouring the ATOMKI claim at $2\sigma$ [774]. Improved independent tests are expected soon.

The state $X$ has been tentatively interpreted as a new physics particle: either a new axion-like pseudo-scalar or an axial vector, coupling to nucleons via a gauge coupling $g \approx 10^{-4}$. The $Xe\bar{e}$ coupling is less constrained, since in this context only the branching ratio $\mathrm{BR}(X \to e^-e^+)$ is relevant. The 2025 results of the $e^-e^+ \to X \to e^-e^+$ PADME experiment show a tentative $\lesssim 2\sigma$ peak at 17 keV, which at present can neither confirm nor exclude the ATOMKI anomaly [775]. Such $X$ could also serve as a mediator between the SM and DM [738]. The axial vector (unlike the vector) can simultaneously fit the three ATOMKI anomalies, in Be, C and He. The pseudo-scalar does not couple to $^{12}$C$^*$, and thus cannot explain that particular excess (unless parity is broken). Other bounds can be satisfied but restrict specific combinations of $X$ couplings to $p$ and $n$ to suspiciously small values: $\pi^0 \to X\gamma$ bounds force a vector $X$ to be proto-phobic, $\pi^+ \to e^+\nu_e X$ bound restrict a pseudo-scalar $X$.

For the ATOMKI anomaly in He several potential explanations within the SM have been put forward as well: that it is a signal of a light tetraquark state; that it is due to higher order effects; or due to a $^4$He resonance with exotic color arrangement in the form of a heavy $(ud)^6$ 'hexadiquark' [738]. These suggestions, however, still need to be validated by calculations from first principles.

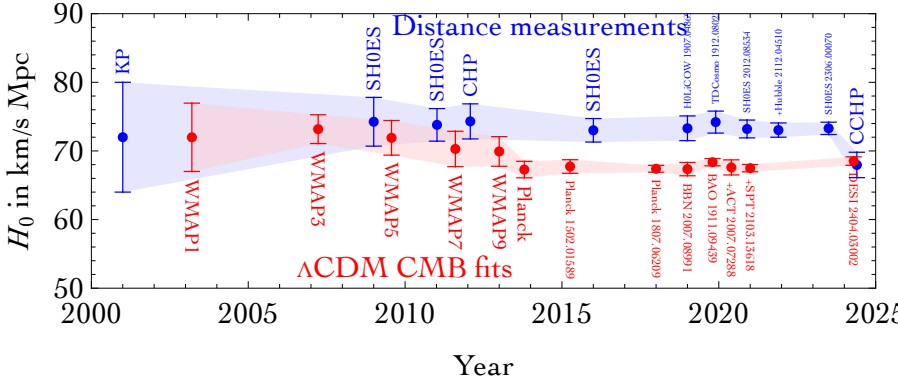

Figure 8.11: *Time series of determinations of the **Hubble constant** $H_0$, either from global $\Lambda CDM$ fits to CMB and structure-formation data (red), or from measurements that use the distance ladders (blue).*

## 8.4 Cosmology: current anomalies

Global analyses fitting cosmological observations to the $\Lambda$CDM model have highlighted potential discrepancies, which might stem from systematic issues. Given the significant role that DM plays in cosmology (see section 1.3), various tentative interpretations involve DM, as reviewed below.

### 8.4.1 The cosmic calibration Hubble tension

Global $\Lambda$CDM fits, in particular the CMB data from PLANCK and at large $\ell \gtrsim 1000$, prefer a present day Hubble constant that is lower than what is measured locally, using cepheids and supernovæ Ia, see fig. 8.11. The benchmark determinations in the two cases come, respectively, from PLANCK, which finds $H_0 = (67.27 \pm 0.6)$ km/s/Mpc, and from the SH0ES collaboration, which finds $H_0 = (73.04 \pm 1.04)$ km/s/Mpc. More generally, the tension is between the values of $H_0$ determined by the 'early Universe' methods (mainly the CMB, but also BBN and BAO) and the 'late Universe' techniques, using supernovæ, the tip of the red giant branch, masers, time delays in strong gravitational lensing (the H0LICOW project), gravitational wave emitters as standard sirens, and a host of other techniques. 2024 results from the Chicago Carnegie Hubble Program (CCHP) fall however in the middle [735]. The magnitude of the tension ranges between $3 - 6\sigma$, depending on how the systematics of the different probes are evaluated and how different measurements are combined. This so called *'Hubble tension'* is one of the most intensely debated issues in cosmology since about 2016 [735].

Significant modifications of cosmology, usually around or before matter/radiation equality and involving unconventional ingredients, are needed to substantially improve the global fit. The first large class of models involves modifying the relative density ($\Omega_{\text{DE}}$) or the equation of state ($w_{\text{DE}}$) of dark energy, because these are degenerate with $H_0$ as inferred from the CMB. The second class of models makes use of the degeneracy between the number of relativistic degrees of freedom, $N_{\text{eff}}$, and $H_0$: one can increase the 'early' value of the Hubble constant and bring it closer to the 'late' one by introducing extra radiation, such as light sterile neutrinos, axions or other light or massless particles.

In this context, DM can play an important role in several different ways [776]. We first review the possibilities within the second class of models introduced above, those in which extra relativistic states are introduced. First, DM can be the origin of the extra radiation: many models have been constructed where (a fraction of) DM is either relativistic or decays into light relativistic degrees of freedom. DM could decay into neutrinos, into photons, into invisible relativistic particles, or into dark mediators that then decay into radiation. DM can also be in the form of PBHs (section 3.1.1) which, if evaporating at the right time, inject relativistic particles into the thermal bath. Second, DM can elastically scatter with

neutrinos, photons, baryons or with some of the dark radiation particles, thereby effectively acquiring (in small part) the behaviour of extra radiation. Third, annihilating DM can inject energy into the thermal bath, thereby keeping higher the fraction of the total energy that is in radiation. This is the case, for instance, in the cannibal DM scenario (section 4.1.9), where $3 \to 2$ processes effectively convert the energy stored in the mass of the eaten particles into thermal energy of the surviving ones.

In the other class of models, where the Hubble tension is alleviated by modifying the properties of dark energy, DM can still play a role through its coupling to DE. This is realized via an effective mechanism that links non-gravitationally the DM and DE energy densities in the cosmological Boltzmann equations (see section 1.3.3). As a result, at specific epochs some of the DM energy density is converted into DE (or vice versa), modifying the cosmological history and therefore the determination of the Hubble constant. There are numerous concrete implementations of this mechanism. Alternatively, there are models in which DM and DE are assumed to be a single fluid, which behaves as DM at early times and as DE at late times. The evolution of the Hubble rate with redshift is therefore modified and this allows to alleviate the tension.

As of this writing it seems that some of the models involving DM can only partially reduce the $H_0$ tension, and that obtaining a convincing solution is not easy. The situation is further complicated by other milder anomalies, as well as by internal tensions, e.g., in Planck data, and tensions between different experiments. The situation may well imply that presently the uncertainties are underestimated, though many experimental checks have been performed.

## 8.4.2  The matter inhomogeneity $S_8$ anomaly

The amount of matter inhomogeneities in the low-redshift Universe $z \lesssim 1$ is quantified by the $S_8 = \sigma_8(\Omega_{\mathrm{m}}/0.3)^{1/2}$ parameter, where $\sigma_8$ is (roughly speaking) the amplitude of the matter power spectrum smoothed over scales of 8 Mpc/$h$.

Several local determinations of $S_8$ performed over the years, e.g. using weak lensing, galaxy clusters and other techniques, had consistently found a value about $3\sigma$ lower than the predictions of global $\Lambda$CDM fits. In other words, the clustering of matter observed in the late Universe seemed smoother than what it would be expected based on the evolution of the early anisotropies (matter clustering at higher redshift, instead, agreed with $\Lambda$CDM). This is the so-called '$S_8$ anomaly'. The situation evolved in 2024 and 2025, when new local determinations, notably by the Kilo-Degree Survey (KiDS) using cosmic shear but also by Act and Spt using CMB lensing (see section 1.2.1), found a value $S_8 \approx 0.82$ in agreement with the global fits [736].

Various authors had tried fitting the $S_8$ anomaly by assuming new physics that slows down the clustering of matter without conflicting with other observables [736]. For instance, by assuming that cold DM decays into warm DM and dark radiation with a lifetime $\tau \approx$ Gyr, or by assuming that DM undergoes DM DM $\leftrightarrow$ DM DM DM scatterings, or that DM interacts with Dark Energy (which is a somewhat natural guess since the anomaly corresponds to the era in which DE starts to dominate over DM).

## 8.4.3  The Desi anomaly

Various experiments measure the late expansion history of the Universe, at scale factor $a \gtrsim 0.1$, by relying on different probes: supernovæ Ia as standard candles, baryon acoustic oscillations as a standard size in matter inhomogeneities, and the CMB. These measurements of $H(a)$ are often presented as $w_{\mathrm{DE}}(a)$, given that the late-time Universe is dominated by Dark Energy and that DE might exhibit an unusual equation of state (we remind that the latter is defined as $p_{\mathrm{DE}} = w_{\mathrm{DE}}\rho_{\mathrm{DE}}$, resulting in $d\ln\rho_{\mathrm{DE}}/d\ln a = -3(1+w_{\mathrm{DE}})$, see Appendix C.2).

The $\Lambda$CDM model assumes a cosmological constant, i.e. a constant $\rho_{\mathrm{DE}}$, and thus $w_{\mathrm{DE}} \equiv -1$. Data from the DESI collaboration, however, indicate a $\approx 3\sigma$ hint for a deviation from $\Lambda$CDM, suggesting that $w_{\mathrm{DE}}$ increases with time, being $w_{\mathrm{DE}} \approx -0.7$ now and $w_{\mathrm{DE}} < -1$ at $a \sim 0.5$ [721]. This is puzzling, since $w_{\mathrm{DE}} < -1$ means that $\rho_{\mathrm{DE}}$ grows with time. A possible way of reconciling a growing $\rho_{\mathrm{DE}}$ with basic principles of physics is that DE interacts with some other component, receiving energy from it. DM is a natural candidate, given that it is the second dominant component in the energy budget of the Universe, while its interaction properties are not known.

A possible theory is that DE is the vacuum energy of some very light scalar field $\phi$ that controls the mass of DM particles $M(\phi)$ [722]. For example DM might be a fermion with a Yukawa coupling to $\phi$. As the DE scalar evolves in the effective potential $V_{\mathrm{eff}} = V(\phi) + \rho_{\mathrm{DM}}$, the DM density $\rho_{\mathrm{DM}} \propto m(\phi(a))/a^3$ can also change, and thus the DM equation of state is also modified. This physics can be presented by including the anomalous variation of DM density in an effective DE energy density

$$\rho_{\mathrm{DE}}^{\mathrm{eff}} = \rho_{\mathrm{DE}} + \rho_{\mathrm{DM}}[1 - M(\phi_0)/M(\phi)], \qquad \Rightarrow \qquad w_{\mathrm{DE}}^{\mathrm{eff}} = \frac{w_{\mathrm{DE}}}{1 + (\rho_{\mathrm{DM}}/\rho_{\mathrm{DE}})[1 - M(\phi_0)/M(\phi)]}, \qquad (8.4)$$

while the effective DM density is the one that would follow from a constant DM mass, $\rho_{\mathrm{DM}}[M(\phi_0)/M(\phi)]$. A DM mass that increases with time can thus mimic a DE with equation of state $w_{\mathrm{DE}}^{\mathrm{eff}} \leq -1$.

In a variation of the same idea, DM is an oscillating axion-like field $a$ with kinetic term $K(\phi)(\partial_\mu a)^2/2$ that depends on $\phi$, that thereby again induces a variation of the axion mass $m_a$ [722].

### 8.4.4 The dipole anomaly

The large observed CMB dipole is interpreted to be due to our local motion with

$$|\boldsymbol{v}_{\mathrm{sun}}| \approx 370\,\mathrm{km/s} \qquad \text{and} \qquad |\boldsymbol{v}_{\mathrm{local\ group}}| \approx 620\,\mathrm{km/s}, \qquad (8.5)$$

relative to the CMB rest frame. According to standard cosmology, this should also be our velocity with respect to distant matter, at red-shifts $z \gtrsim 1$. The same dipole in the observed sky brightness should thus also be seen with astrophysical probes. However, some (but not all, see J. Darling in [733]) surveys find a velocity that is a factor of a few larger, in roughly the same direction [733], where one should keep in mind that these astrophysical measurements of the dipole are more difficult than the measurement of the CMB dipole. If confirmed, this would challenge the view that matter is nearly homogeneously distributed on large scales. Standard cosmology would then need a drastic revision, with implications for DM.

### 8.4.5 The EDGES 21 cm anomaly

In 2018, the Experiment to Detect the Global Epoch of reionization Signature (EDGES) reported an observation of the signature corresponding to the absorption of 21cm radiation by cosmic neutral hydrogen at redshift $z \approx 17$. The amount of absorption, in its simplest interpretation, implied that the temperature of the neutral gas at that epoch was significantly lower than what is expected in the standard $\Lambda$CDM cosmology, with a statistical significance of about $3.8\sigma$ [777].

This was surprising given that the thermal history of cosmic hydrogen, while rather complex, is well understood within the standard cosmology.[12] It is worth reviewing qualitatively the main phases of the evolution in order to better understand the possible origins of the anomaly. Before recombination, the photon and the gas fluids had the same temperature. At the moment of recombination ($z \sim 1100$, i.e., when the Universe was 380 kyr old), the photons, which from that moment constitute the CMB, and the newly formed neutral hydrogen atoms scattered for the last time. However, the residual Compton interactions kept the two fluids thermally coupled for much longer, until approximately $z \sim 150$ (or 8

---

[12]The broad formalism for the evolution of the gas temperature has already been sketched in section 6.11.3.

Myr). Subsequently, the gas, also referred to as the intergalactic medium (IGM), thermally decoupled from the CMB and began cooling adiabatically as $T_{igm} \propto (1+z)^2$. This is faster than the cooling of CMB, whose temperature scales as $T_{CMB} \propto (1+z)$. As a result, $T_{igm}(z)$ became lower than $T_{CMB}(z)$, starting from their common value at $z \sim 150$. This remained true at least until the era of cosmic dawn, when the first stars started emitting light, eventually raising the gas temperature above that of the CMB at $z \lesssim 15$. Distinct from $T_{igm}$, the hydrogen 'spin temperature' $T_S$ is defined by $n_1/n_0 \equiv 3e^{-\Delta E/T_S}$ and parameterizes the relative population of the two ground state levels of cosmic neutral hydrogen, with spin $S = 0$ or $S = 1$ and the energy difference $\Delta E = 2\pi/(21\,\text{cm})$. During the dark ages ($z \sim 1100 \to 20$) and through cosmic dawn ($z \sim 20 \to 15$), $T_S$ evolved between the 'floor' value $T_{igm}$ and the 'ceiling' value $T_{CMB}$. Whenever $T_S < T_{CMB}$, the background CMB photons shining on the clouds of cold hydrogen gas get absorbed, generating a dip in the spectrum. This is notably the case around $z \approx 17$ leading to the expectation that CMB photons that had a wavelength of 21 cm at that time, corresponding to 70-80 MHz today, i.e., the very low-energy portion of the CMB, should be missing from the measured spectrum of the CMB. By analyzing the depth of this absorption dip, one can obtain information about the temperature of the gas and the CMB at that epoch. This dip in the CMB spectrum is exactly what EDGES measured.

More precisely, the experiment measured the quantity $T_{21}(z) \approx 23\,\text{mK}\,(1 - T_{CMB}(z)/T_S(z))$, which expresses the difference between the brightness temperature of the 21 cm hydrogen emission and the temperature of the low-energy portion of the CMB radiation. EDGES found $T_{21} \approx -0.5^{+0.2}_{-0.5}\,\text{K}$ at 99% C.L., while standard astrophysics predicts $T_{21} \approx -0.2\,\text{K}$ at $z \approx 17$. Qualitatively, this means that the difference between the floor ($T_{igm}$) and the ceiling ($T_{CMB}$), between which $T_S$ can evolve, is larger than expected.

The EDGES measurement required a careful subtraction of an overwhelmingly intense foreground, approximately $10^4$ times stronger than the signal under investigation. Assuming that the net result is correct, the detected anomalous temperature difference requires physics beyond the Standard Model. There are two ways to solve the discrepancy with new physics. On one hand, one can 'lower the floor', i.e., cool the hydrogen gas more efficiently than in the standard cosmology. On the other hand, one can 'raise the ceiling', i.e., increase the temperature of the CMB photons.

DM can potentially play a pivotal role in both of these scenarios [730]. On one side, since DM decoupled much earlier than the gas, the DM fluid is much colder than hydrogen during the dark ages, including at $z \approx 17$. Prompted by the EDGES measurement, it was thus quickly suggested that DM-baryon scatterings could offer an efficient mechanism to cool down the hydrogen gas. These scatterings might also occur between baryons and a fraction of DM, possibly with DM being milli-charged. On the other hand, additional 21-cm radiation could be injected via the decays or annihilations of DM particles, from the evaporation of Primordial BH DM, or via the conversion of axion-like DM into photons. This would effectively heat the CMB spectrum. Observationally this is not a problem, since the CMB is poorly measured in its very low-energy portion.

The EDGES results can be approached from a different angle: one can give up on trying to explain the discrepancy, and simply focus on the fact that an absorption feature has been detected. This implies that the gas was colder than the CMB at $z \approx 17$, which then imposes stringent bounds on DM annihilations or decays that would have heat the gas [672].

Future experiments such as LOFAR, SARAS, BIGHORNS, LEDA, MWA, ASSASSIN or SKA [777] will test the absorption signal and may shed light on the potential involvement of DM. Preliminary results from the SARAS3 experiment in 2021 (Singh et al. (2022) in [777]) do not confirm the EDGES findings.

## 8.5 Astrophysics: current anomalies

An astrophysical anomaly may sound like an oxymoron,[13] given that in astrophysics one cannot perform repeatable experiments, but rather only make observations, contrary to fundamental physics that can be done in a laboratory. Taking into account this limitation, we summarize below possible issues discussed in the literature.

First, we discuss various puzzling features related to the DM density distribution on galactic (section 8.5.1) and sub-galactic (section 8.5.2) scales. These are also referred to, rather dramatically, as the 'small scale crisis' [778]. It is worth noticing that these features often point to a disagreement between numerical simulations (see section 1.3.2) and observations, i.e., they point to our still limited understanding of what DM is and how it behaves. This is contrary to most of the other anomalies discussed in the other sections of this chapter, which instead are deviations from known physics and thus hint at DM. Then, we discuss the black hole mass gap (section 8.5.3) and stochastic gravitational waves (section 8.5.4).

## 8.5.1   Galactic-scale anomalies

▶ The "**cusp-core problem**" [747]. Numerical $N$-body simulations that include only collision-less cold DM and gravity predict DM halo density profiles that for $r \to 0$ grow as $1/r^\gamma$, $\gamma \approx 1 - 1.5$, giving a cusp in the galactic center. Observations, however, tend to favour constant density cores, see section 2.2.1. Various authors speculated that this so called *cusp-core problem*[14] points to a more complex DM. Cored profiles can, for instance, arise if DM is ultralight (fuzzy DM, section 3.4), with the de Broglie wavelength comparable to the size of the DM halo core. Another possibility is that DM is self-interacting (section 3.3.3), in which case the interactions affect the evolution of DM halos. Initially, warm DM (section 3.3.1) was also invoked as a possible solution, but this tends to run afoul of other observations, which bound the DM mass to be above few keV, so that DM needs to be effectively cold as far as the cusp-core problem is concerned.

The inclusion of baryons in the $N$-body simulations at least partially resolves the cusp-core problem. Actually, early simulations that included baryons showed the opposite effect, where the inclusion of baryons made the cusp-core problem worse since baryons can pull more DM into the centre as the gas cools and condenses, in the so called *adiabatic contraction* process (see, e.g., Gnedin et al. (2011) [747]). This was later shown to be due to a too coarse spatial resolution in the early simulations. Some early simulations also indicated that the formation of a central bar in the galaxies could turn a cusp into a relatively large core within a few orbital times. This was later shown to be due to too simplified numerical assumptions. The situation changed around 2013 when hydro-dynamical simulations became more refined. In particular, the works of Di Cintio et al. (2014), Tollet et al. (2016), Read et al. (2016) and Katz et al. (2016) in [747] showed that whether or not a cusp forms at the center of a simulated galaxy depends on the stellar-to-halo mass ratio, and can also be a function of time. In the formation of galaxies with few stars, the injection of energy from the stellar feedback is insufficient to significantly modify the inner density of the DM halo. Such galaxies therefore retain the cuspy profile which is typical of DM-only simulations. At higher stellar-to-halo mass ratios ($\mathcal{O}(10^{-3} - 10^{-2})$), the stellar activity (in particular the supernovae explosions) 'pushes out' DM, which makes the gravitational potential shallower and therefore drives the creation of a cored profile. At even higher stellar-to-halo mass ratios ($\gtrsim$ few $10^{-2}$), the abundance of stars at the center of the galaxy instead deepens the gravitational potential, recreating a cuspy profile. The stellar-to-halo mass ratio can evolve with time. The inner density profile of a galaxy can thus change substantially as the galaxy grows in both the stellar and total mass. In the case of the

---

[13] "*A single hydrogen atom in conflict with quantum mechanics would be a tragedy. A single galaxy in conflict with $\Lambda$CDM can be an outlier*" ($\sim$ M. Bauer).

[14] From rotation curves it is easier to measure the total mass up to a certain radius in the inner region, while its derivative, i.e., the DM density profile, is harder to determine. The problem is thus more robustly phrased as the *"inner mass deficit"*: DM only simulations predict more DM in the inner regions of galaxies than what is observed.

Milky Way, adopting the estimates given in section 2.2.2, the stellar-to-halo mass ratio is quite high ($\approx 0.08$) and thus points towards a cusped inner profile.[15]

▶ The "**diversity problem**" [748]. Spiral galaxies with the same maximal circular velocity show larger spread in their interiors and their core densities (Kuzio et al. (2009) and Oman et al. (2015) in [748]) than what is expected from the cold DM $N$-body simulations (Navarro et al. (1996) and Bullock et al. (1999) in [748]). While for the simulated galaxies the shape of the rotation curves varies as a function of galactic mass, there is little variation between different simulated galaxies that have the same maximal circular velocity. That is, in numerical simulations two galaxies with similar halo masses will have comparable total stellar mass, mass concentration and rotation curves, while observations show a diverse range for these quantities.

The possible resolutions are the same as for the cusp-core problem: (i) DM dynamics could be more complex, (ii) current simulations may fail to reproduce correctly the effect of baryons, or (iii) the observational determination of mass profiles in the inner parts of the galaxies could be incorrect. However, at present it is far from clear which of these resolutions is the correct one.

▶ The problem of **DM dynamical friction** [779]. As discussed in section 2.5, the gravitational pull from DM produces a frictional force on celestial bodies moving through regions with high DM density. Several studies, however, suggest that certain astrophysical systems do not exhibit this effect. For example, some studies have argued that the central bars, present in many of the observed disc galaxies, should have experienced more slowing down from the DM halo friction than what is observed. Similarly, it was argued that the globular clusters in the Fornax dwarf galaxy should have sunk in the center more rapidly than what the observations indicate (this is the so called *timing problem*: why are at least five of these clusters observed at significant distances from the center of Fornax, if they should have gravitated inward long ago?). This has been used to argue *against* the very existence of DM halos (see, e.g., M. Roshan et al. [77]), or at the very least as an evidence for significant tension with the predictions of the ΛCDM model.

However, more mundane astrophysical explanations do exist. Some simulations suggest that rotating bars within DM halos can persist, if the bars formed late in the galaxy's history, potentially through a merging event. Additionally, certain studies find that a cored DM distribution in Fornax could prevent the inward migration of globular clusters, a phenomenon referred to as *core stalling* [77].

▶ The **baryonic Tully-Fisher relation** [780] is an experimentally observed relation between the amount of baryonic matter in the galaxy and the maximal rotational velocity in the galaxy (measuring the total amount of DM and visible mass in the galaxy).

While such a relation between baryon content of galaxies and their halo masses is believed to arise naturally from galaxy formation, it is currently still hard to tell whether the theoretical predictions from ΛCDM agree with observations in detail. For further discussion of this issue, see section 11.2.2.

▶ Starting from 2022, the James Webb Space Telescope (JWST) observed **early galaxies** at redshift $z \gtrsim 7$. These early galaxies are seemingly too massive to be present in such an early cosmological epoch, at least according to the standard ΛCDM cosmology, complemented by astrophysical models of star formation [781]. Various more efficient astrophysical star formation processes could explain such observations. Alternatively, observations might point to new DM physics, such as a higher

---

[15]As a word of caution, we mention, however, that the studies quoted above in [747] draw their conclusions using galaxies somewhat smaller than the MW, and that the slope of the DM density is measured within a rather restricted range of distances from the center, typically $1-2\%$ of the virial radius.

DM power spectrum at the relevant scales. Some authors instead interpret this possible anomaly as evidence for the MOND alternative to DM [781] (to be discussed in chapter 11).

## 8.5.2  Subgalactic-scale anomalies

▶ The **missing satellite problem** [749]. The DM-only $N$-body simulations predict many more DM sub-halos ('satellites') than the observed number of 'classical' dwarf galaxies, known prior to the improved searches that started in mid 2000s (see section 2.2.3). This discrepancy was dubbed the *missing satellite problem*, and is now largely resolved.

First of all, many more new faint satellites were discovered in the Milky Way and Andromeda (M31) galaxies. Furthermore, the $N$-body simulations have improved: they have a better resolution, they include baryons and they can now resolve previously neglected processes such as the fact that the central galaxy potential destroys subhalos on orbits with small pericentres. As a result, the state-of-the art $\Lambda$CDM hydrodynamical numerical simulations are now essentially consistent with the observed number of satellite galaxies, at least once the corrections for the detection efficiencies of the surveys are included. In summary, simulations and observations met mid-way: satellites form in lower numbers then initially predicted, but many are too dark to be seen.

▶ **"Too big to fail"** [743]. The most massive observed satellites of the Milky Way are smaller than the most massive subhalos obtained in DM-only $N$-body simulations.[16] The question therefore arises: if the Milky Way is supposed to have very massive satellites, where are they and why don't we see them? One possibility would be that such very massive satellites simply do not contain stars, are thus completely dark, and have not been observed. This would be very surprising since such subhalos are massive enough that their gas should have cooled and formed stars, like it did in the smaller subhalos. In other words, the very massive subhalos are *"too big to fail" (forming stars)*.

A more likely resolution therefore lies elsewhere. One option is that the total mass of the Milky Way halo is overestimated — if the mass of Milky Way is in the lower part of the currently allowed observational range (see eq. (2.13)) this would imply that the most massive satellites are also expected to be lighter, in line with observations. The other possibility is simply statistics - there is large halo-to-halo scatter for the mass of the most massive subhalos, and Milky Way could just happen to be one of the downward fluctuations. The too big to fail problem is also less of an issue when comparing with $N$-body simulations that include baryons, in particular because in that case the galaxies can form DM cores instead of cuspy profiles (see section 8.5.1), and thus lead to smaller halo masses. In summary, there appears to be consensus among current cosmological simulations that there is no 'too big to fail' problem for Milky Way and M31 satellites anymore, while it may still be present for isolated dwarf galaxies in the nearby Universe.

▶ **Planes of satellites** [750]. A large fraction of the Milky Way satellites lie in a thin planar disk almost perpendicular to the galactic disk, which appears to be corotating. Something similar is also observed for around half of the satellites of the Andromeda galaxy M31. And some nearby galaxies also show such planar distributions of satellites. This has been one of the more persistent discrepancies between simulations and observations, since simulations predict satellites to be distributed roughly isotropically (spherically or at most slightly prolate), and to possess random motions.

---

[16]The same discrepancy is found also for Andromeda (M31) and other nearby galaxies.

The resolution is not clear, though several possibilities have been discussed:[17] (i) that the presence of massive satellites, such as the large Magellanic cloud, boosts the planarity of the satellite distribution (this is seen in the simulations); (ii) it is possible that the surrounding larger scale cosmological structure needs to be included in the simulations (this is hinted at by the fact that the planar structures of dwarf galaxies in the whole local group of galaxies shows some degree of alignment; by the way, this is claimed to work also in modified Newtonian dynamics, without DM, see Bílek et al. (2017) [750]); (iii) the observed alignment could also be just temporary, a situation that appears to be quite likely in simulations (plane configurations are found to last $\sim 500$ Myr).

▶ **Quiescent fractions** [782]. Nearly all observed isolated (non-satellite) dwarf galaxies are star-forming, while nearly all satellites of the Milky Way and M31 are *quiescent*, with no gas and no star formation. This seems to be in contrast to other Milky Way like galaxies in our local group, for which nearly all their satellites are also star-forming.

The resolution likely has little to do with DM properties. However, it does raise the following question: if the Milky Way and M31 are outliers in this respect, how representatives are their other properties of an average galaxy, which is what simulations aim to reproduce?

### 8.5.3   Black hole mass gaps

In 2019, the ADVANCED LIGO and VIRGO collaborations detected a gravitational wave signal from a merger of two black holes, with masses $\approx 66\,M_\odot$ and $\approx 85\,M_\odot$ (later revised to slightly higher values). There is about $3\sigma$ probability that at least one of the two BHs lies in the so-called 'BH upper mass gap' or the 'pair-instability gap', i.e., in the range $50\,M_\odot \lesssim M \lesssim 120\,M_\odot$. Subsequent results by the LIGO-VIRGO-KAGRA collaboration added a few more candidates with similar characteristics [783].

These results are puzzling — normal stellar evolution predicts that there should be no BHs with masses in this range, if they were produced from the collapse of a star, due to the following considerations. In the cores of stars with masses $\gtrsim 50 M_\odot$ the density and temperature is high enough that electron-positron pairs can be efficiently produced from the plasma, which then decreases the photon pressure. Roughly speaking, the energy required to produce the $e^\pm$ rest mass comes at the expense of the thermal support, causing the star to become unstable and undergo a thermonuclear explosion, resulting in no black hole remnant. These catastrophic events go by the name of pair-instability supernovæ [784].[18] In sufficiently heavy objects, $M \gtrsim 120 M_\odot$, the pair-creating instability is quenched since the energy from the collapse/contraction is large enough to disintegrate heavy elements, reaching an equilibrium between the two processes, allowing again for the formation of a BH remnant. In summary, in the $50\,M_\odot \lesssim M \lesssim 120\,M_\odot$ mass range no standard astrophysical BHs should exist.[19]

There are two possible ways that DM could help resolve the BH mass gap anomaly [728]. First, the BHs within the mass gap could be of a primordial origin: primordial BHs can be created with an arbitrary mass, and would constitute all or part of the DM. The usual constraints, discussed in section 3.1.1, apply to this scenario. These imply that if all such primordial BHs are of the same mass, they cannot explain all of the DM abundance, see fig. 3.3, while primordial BHs with a more spread out mass distribution could.

---

[17]The trivial possibility that this is due to a selection bias, i.e., only the satellites sticking out of the disk are seen, can be quickly dismissed: the galactic disk of the Milky Way only obscures two relatively thin wedges of about 24 degrees, and outer galaxies are observed even more clearly.

[18]There is also a related variant, the pulsational pair-instability supernovæ.

[19]Nonetheless, some astrophysical scenarios to populate the mass gap do exist. For example, BHs within the mass gap could have formed from mergers of lighter BHs (the hierarchical scenario), from very massive stars, or through accretion of matter. The rates for these processes depend on the astrophysical environments in which the BHs are created, and are thus fraught with uncertainties.

The second possibility is that the physics of very massive population III stars could be modified by an extra cooling channel from couplings of photons or electrons to new light dark particles. In the literature, the scenarios of annihilations into DM, the presence of new light particles such as an axion or a dark photon, or a possibility of a neutrino with a sizable neutrino magnetic moment, were considered. However, it is not clear whether any of these cases leads to a large enough increase in the mass of the BH remnant, in order to fully explain the detected events.

There is also a 'lower mass gap' in the range between $3M_\odot$ (the presumed highest mass for a neutron star) and $5M_\odot$ (the presumed lowest-mass black hole that can form from stellar core collapse), which should be empty of compact objects. In 2024, however, the LIGO/VIRGO/KAGRA collaborations detected a gravitational wave event from a merger of a neutron star with a likely black hole of mass $M = (2.5 - 4.5)M_\odot$, thus falling in the purported lower mass gap. This anomalously light BH could originate from a main sequence star that accumulated DM until forming a BH [785], as discussed in section 6.10.

## 8.5.4 The gravitational waves detected by Pulsar Time Array

A pulsar timing array (PTA) is a name for a network of millisecond pulsars that is being continuously monitored. PTA is in its essence a galactic size detector composed of very stable clocks, which can then be used to search for very low frequency signals that would imprint themselves as correlations among the shifts in arrival times of light pulses emitted by the pulsars. These can get shifted by various mundane effects, but would, more interestingly, also get shifted by background Gravitational Waves (GW). After monitoring for more than a decade about 100 stable pulsars, the NANOGRAV collaboration in 2020, PPTA and EPTA in 2021, and the joint IPTA analysis in 2023, reported an observation of a correlation between distant pulsars, which could possibly be due to a stochastic GW background.

In 2023 NANOGRAV and EPTA [98] furthermore reported a $\sim 4\sigma$ evidence for the characteristic feature of spin 2 gravitational waves: quadrupolar spatial correlations in the part of the GW signal closer to us, and thereby common over the PTA [786]. GWs have been observed roughly in the $(1 - 10)$ nHz frequency range, set by the inverse of the observation time. Longer monitoring times will allow to probe lower frequencies. Parameterizing the GW spectrum of density $\rho_{\rm GW}$ with a power law ansatz, the fraction of cosmological energy density that is in GWs is then given by,

$$\Omega_{\rm GW} \equiv \frac{1}{\rho_{\rm cr}}\frac{d\rho_{\rm GW}}{d\ln f} = \frac{\pi f^2 h_c^2}{4G}, \qquad \text{where} \quad h_c(f) = A_{\rm GW}\left(\frac{f}{f_{\rm PTA}}\right)^\beta. \tag{8.6}$$

The spectral slope $\beta \approx -0.1 \pm 0.3$ and the amplitude $\Omega_{\rm GW}h^2 \sim 10^{-9-10}$ measured at the frequency $f_{\rm PTA} \equiv 1/10$ yr roughly agree between different experiments and different analyses. They are roughly compatible with the astrophysical background, expected to be generated by a population of in-spiralling super-massive black hole binaries (SMBH) with masses $M_{1,2} \sim 10^{8-9}M_\odot$ formed via galaxy mergers [787]. Each individual source contributes as $h_c \propto f^{-2/3}$ and thereby $\Omega_{\rm GW} \propto f^{2/3}$ in the limit where it undergoes circular non-relativistic orbits with energy losses dominated by the gravitational waves, $W_{\rm GW} = 32G\mu^2\omega^6 r^4/5$, where $\mu = M_1 M_2/(M_1 + M_2)$ is the reduced mass of a two-body system and $\omega = \pi f$. The spectral power $\beta = -2/3$ follows from $dE_{\rm GW}/d\omega = W_{\rm GW}/d\omega/dt$, after using energy conservation $d(\mu v^2/2 - GM_1M_2/r) = -W_{\rm GW}$ to compute $d\omega/dt$ and the Newton force $v^2/r = \omega^2 r = G(M_1 + M_2)/r^2$ to eliminate $r$ in favour of $\omega$. If these GWs are really due to SMBH, future data are expected to see an anisotropy and individual sources.

If the PTA signal of GWs is due to merging SMBH binary systems, this will allow to probe the DM density $\rho = \rho_b + \rho_{\rm DM}$ around the SMBH via dynamical friction mechanism and the resulting extra energy loss (see section 2.5). The energy loss due to DM dynamical friction, $W_{\rm DM}$ in eq. (2.43), can be comparable to the loss due to the emissions of gravitation waves, $W_{\rm GW}$, in the observed nHz frequency range. The presence of DM can therefore affect the spectral slope $\beta > -2/3$, and the phases of PTA

pulses. Maybe precise measurements of GW could tell something about the sum of matter and DM density [787].

Various authors also considered the possibility that the GW signal in PTAs could be due to new physics instead. Ignoring the signals that are not quadrupolar, and focusing on those possibly related to DM, the proposed interpretations [788] can be summarized as:

○ A **phase transition** that happened after the big bang at temperature $T$ could produce GWs peaked around frequency $f \sim T T_0/M_{\rm Pl}$, corresponding to the Hubble rate at temperature $T$, red-shifted by $T_0/T$. To explain the PTA signal one therefore needs a phase transition at $T \sim (10-100)\,{\rm MeV}$. The phase transition has to be very strongly first order, given that the nearby Hubble patches colliding relativistically would generate $\Omega_{\rm GW} \sim 1$, while deviations from this optimal limit generate a smaller signal, $\Omega_{\rm GW} \sim \alpha^2 v^3 (H/\beta)^2$, where $\alpha < 1$ is the fraction of the energy involved, $v < 1$ is the collision velocity, and $1/\beta$ is the time-scale of the transition (a fast transition leads to many small bubbles). Nothing like this happens in the Standard Model; new physics is needed. Due to experimental and BBN constraints on new particles with MeV-scale masses the phase transition needs to occur in a 'dark' sector feebly coupled to the SM, with possible connections to DM.

○ **Primordial inhomogeneities** with an enhanced power spectrum $P_\zeta(k) \gg 10^{-9}$ could possibly arise during the later phases of inflation (theories will be discussed in section 4.6.1) and would generate gravitational waves on sub-horizon scales with frequency spectrum $\Omega_{\rm GW}(f) \sim 10^{-5} P_\zeta^2(2\pi f)$ and also Primordial Black Holes with abundance $\Omega_{\rm PBH}(M) \sim e^{-10^{-2}/P_\zeta(k)}$. The PBH mass is roughly given by the horizon mass when the mode $k$ re-enters the horizon at temperature $T \sim k T_0/H_0$, so $M \sim \rho V \sim T^4/H^3 \sim M_{\rm Pl}^3/T^2$. Adding order unity factors, this means that the nHz gravitational waves claimed by PTA would form at $T \sim (10-100)\,{\rm MeV}$ and have a mass distribution peaked around $M_\odot$. Such PBHs cannot be DM, since only PBH with mass distribution peaked in the range $10^{-11-15}M_\odot$ can comprise all of DM, see section 3.1.1. The GW amplitude claimed by PTA corresponds to $\max_k P_\zeta(k) \sim 10^{-2}$, and imply a PBH abundance comparable to current bounds (Franciolini et al. in [788] claim a mild contradiction). The GW spectrum might extend to larger frequencies $f$, to be probed e.g. by the planned LISA mission. The PBH mergers would also contribute to GWs events, with a possible relation to the anomaly of section 8.5.3.

○ **Cosmic Strings**, possibly generated after a symmetry breaking that could have happened in the early Universe, would oscillate and emit GWs with a characteristic spectrum set by the string tension $\mu$. The PTA signal can be fit for $G\mu \sim 10^{-10}$ (stable strings) or $G\mu \sim 10^{-6}$ (metastable strings). The same parameter $G\mu$ controls the cosmic string density, with $\Omega_{\rm CS} \propto G\mu$ in the simplest scaling regime. However, it is not easy to reproduce the GW spectrum favoured by the PTA data without conflicting with the bounds at higher GW frequencies $f$, even allowing for a low inter-commutation string probability of $\sim 10^{-2}$. In this case the stochastic GWs could also be observed by Advanced LIGO in a relatively near future.

○ Alternatively, GWs could be sourced by quantum fluctuations of a **dark photon** field $X$ that was created during the rolling of an axion-like scalar $a$. Assuming the interaction $-qaX_{\mu\nu}\tilde{X}^{\mu\nu}/4f_a$, the amplitude and the frequency of the observed PTA signal point to an axion mass $m_a \approx 10^{-13}$ eV and axion-like decay constant $f_a \approx 5\,10^{17}$ GeV (for $q \approx 50$).

Finally, the PTA data also allow to constrain **ultra-light DM** with mass $M \sim 10^{-23}\,{\rm eV}$. If such a light DM does not interact with the SM, DM density fluctuations would give metric fluctuations producing a (non quadrupolar) signal correlated among distant pulsars. A DM coupled to the SM instead induces tiny oscillations of the SM parameters, e.g., of $\Lambda_{\rm QCD}$. As a result, the moments of inertia of pulsars would also oscillate, making their frequencies change in order to conserve angular momentum. The same physics would also affect our clocks, see section 5.8.

# 8.6 Anomalous anomalies

Sometimes interpreting anomalous events in terms of Dark Matter can lead to some eyebrow-raising hypotheses, such as:

∅ Randall and Riece (2014) [71] speculated that a dense disk along the galactic plane made out of a sub-component of dissipative Dark Matter triggered comet impacts that might have caused mass extinctions of life on Earth with possible $\approx 35$ Myr periodicity (including the Cretaceous−Paleogene **extinction of dinosaurs** $\approx 66$ Myr ago). The suggested mechanism is that since the Sun, as it orbits the Galactic Center, oscillates above and below the galactic plane with a period of about 35 Myr, it would periodically traverse the dark disk; the large density gradient during the crossing of the thin disk would exert a gravitational perturbation on the Oort cloud, which then triggers an enhanced meteor shower on Earth. This possibility is now excluded, see section 2.4.3. Similar mechanisms were earlier proposed without invoking DM.

∅ Froggatt and Nielsen (2014) [789] speculated that the **Tunguska explosion** in 1908 was due to a chunk of DM composed out of quarks (see section 9.1.4), with a total mass $M \sim 10^8$ kg, radius $R \sim$ cm, and velocity $v \sim 200$ km/s, which hit the Earth, penetrated it, released its energy, and possibly stimulated volcanic activity. The Tunguska explosion is usually attributed to an asteroid or a comet that disintegrated in the air, since a crater was never unambiguously identified.

∅ Starkman et al. (2022) [99] speculated that four unusually **straight lightening bolts** filmed in Australia might have been due to four macroscopic DM objects that crossed the Earth atmosphere in the presence of an electric field. Their paper was retracted when the anomaly was later understood as an (anomalous) effect in the camera.

# Part III

# Dark Matter theories

# Theory Introduction

The ultimate goal of searches for DM particles and their interactions is to establish the correct theory of DM. If successful, this will likely reduce the updated version of the present review to an addition of an extra line in the expression for the SM Lagrangian, as well as some accompanying discussion of the resulting phenomenology.

At present, however, we have only limited knowledge about DM, and no definitive experimental signal of its interactions. As a consequence, there is a proliferation of theoretical ideas about what DM could be. In fact, in its essence DM is such a simple and general idea that proposing specific realizations may even be *too easy*. From the perspective of particle physics, it is reasonable to expect that DM will be described by a renormalizable relativistic Quantum Field Theory (QFT), a highly predictive approach that requires only the symmetries of the Lagrangian and the quantum numbers of the DM particle(s) to make predictions, using a limited set of parameters. The triumph of this approach is the Standard Model, which enabled particle physicists to anticipate numerous experimental findings. In parts of the DM research field, especially when focusing on cosmology or astrophysics, the bar for writing down theories of DM is lower, focusing on phenomenological consequences at hand and only hoping that the effective description fits into a QFT framework.

This leaves us with a daunting task of reviewing the theories of DM, where on one hand we run the risk of producing a long useless list of possibilities, and on the other hand we may be leaving out the theoretical description of DM that turns out to be realized in Nature. In this endeavour, we will use the crutch that many theorists lean on when trying to compensate for the lack of experimental information about DM, and focus more heavily on speculations that are *theoretically well-motivated*. The interpretation of what is *theoretically well-motivated* is subjective, but it usually implies one of the following:

1. **DM theories motivated by other issues**, such as the naturalness of the Higgs mass, the QCD $\theta$ problem, the nature of dark energy, etc.

2. **Simple theories**, where this either denotes models that add the fewest additional degrees of freedom (a single scalar or fermion) or possess a minimal structure (one electroweak multiplet, a single new symmetry, only gravitational interactions, etc). Note that sectors with complex phenomenology can sometimes follow from simple theories (an example is, for instance, the SU(3) strong interaction in the SM).

3. **Predictive theories**, where DM properties are predicted in terms of a small number of free parameters, ideally zero.

4. **Elegant theories**, which provide a plausible explanation for some of the puzzling DM properties, such as: why is DM stable? Why is DM neutral? Why are the abundances of DM and visible matter comparable?

5. **Prototypical theories**. Theories that employ ingredients that tend to appear, at least in some limit, in more complex theories. For example DM as a supersymmetric wino can behave as any other fermionic electroweak triplet with zero hypercharge.

6. **Theories that predict novel signals**, or at least lead to very distinct signals.

Fig. 9.1 shows a map of some DM theories that score high on one or more of the above criteria, many of which we, at least briefly, review below.

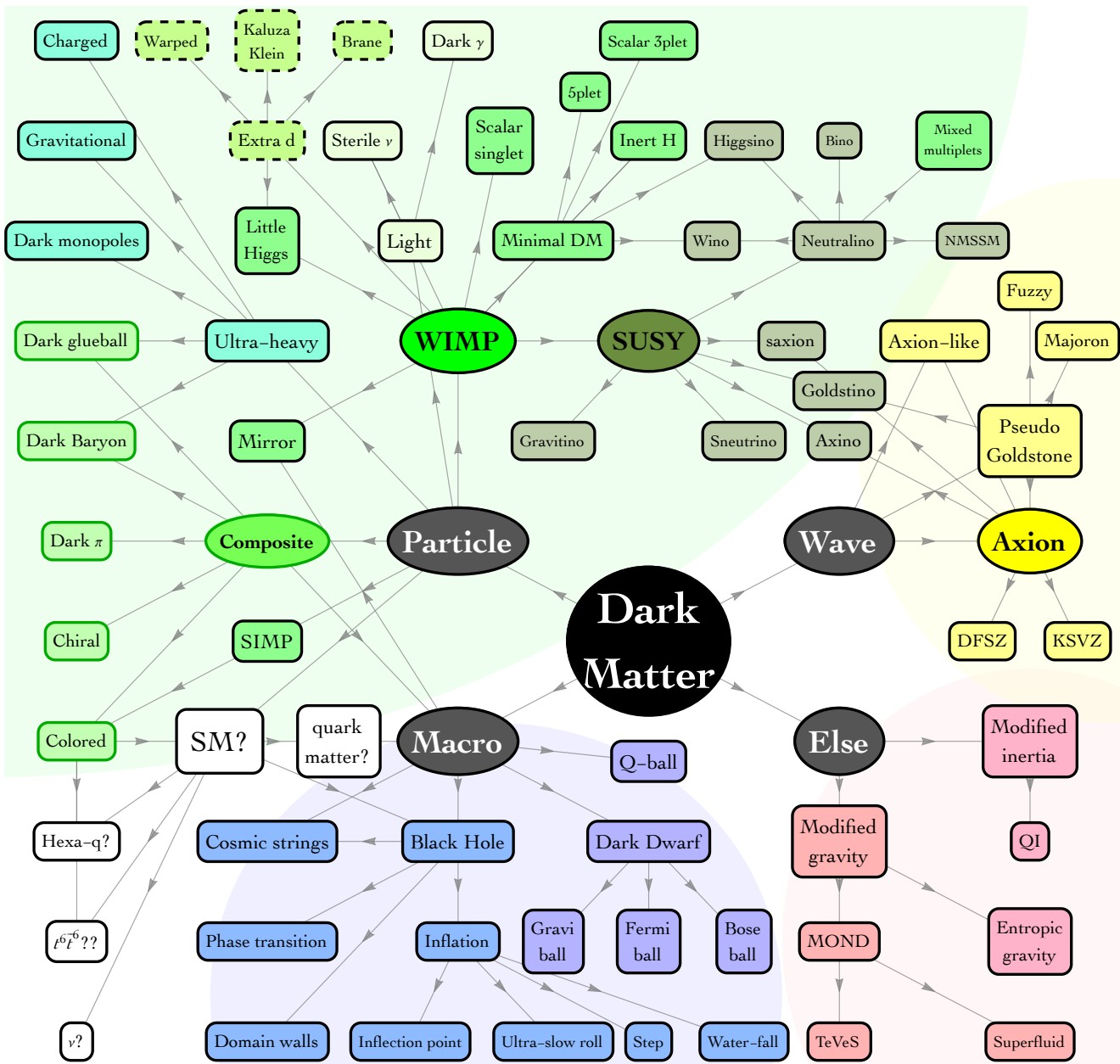

Figure 9.1: *A map showcasing some of the main tentative DM theories.*

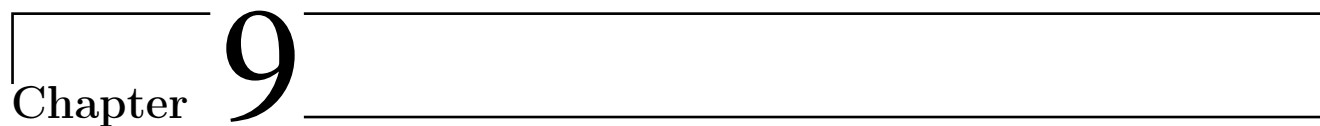

# Chapter 9

# Theories of Dark Matter

In this chapter we present varios speculative theories of particle DM that primarily align with criteria 2 to 6 listed on page 309, i.e., the simple, predictive, elegant, prototypical and/or novel theories that are usually *not* motivated by external particle physics considerations. The subsequent chapter 10 will focus instead on theories that mostly align with criterion 1, i.e., the theories motivated by 'bigger' particle physics issues.

The chapter is organized, somewhat arbitrarily, as follows. In section 9.1 we first summarize why the Standard Model fails to provide a good DM candidate. We then explore DM candidates beyond the SM, using their SM interactions as an organizing principle. First, we consider DM candidates that do not carry any SM charge: a scalar singlet (section 9.2.1) or a fermionic singlet (section 9.2.2). Next, we move to DM candidates that are charged under the SM gauge group: DM might carry a small nonzero electric charge (section 9.3.1, with a more detailed discussion already given in 3.3.2) or be charged under QCD (section 9.3.2), while in section 9.3.3 we discuss a class of DM particles that carry a weak charge (WIMPs) and its extensions. The Minimal Dark Matter candidates (section 9.3.4) are special cases within the WIMP category. This is followed by a review of DM particles that are charged under hypothetical new forces or that interact with the SM via a 'portal' (section 9.4). The new forces can allow DM to be a composite particle, as discussed in section 9.5. Dark sectors with a non-trivial group structure can produce DM that consists of dark monopoles, as we discuss in section 9.6. Some dark sectors can generate a DM asymmetry analogous to the matter asymmetry, as discussed in section 9.7. Finally, in section 9.8, we consider the possibility that DM has no other interactions but the gravitational one.

## 9.1  Dark Matter in the Standard Model?

Table 9.1 summarizes the particle content of the Standard Model (SM). Given this content, we first discuss why there is no dark matter candidate within the SM, summarising the non-trivial attempts in this direction. We first discuss elementary particles, moving then to bound composite states bound by QCD, electroweak forces, nuclear forces, gravitational forces.

### 9.1.1  Elementary: neutrino DM?

SM neutrinos are electrically neutral, weakly-interacting, stable particles, endowed with a non-zero mass: they pass most of the criteria discussed in section 3.3 for particle DM, and thus are at first sight a good candidate.

In addition, neutrinos are very abundant in the Universe. According to Big Bang cosmology, the SM

| Fields | spin | $U(1)_Y$ | $SU(2)_L$ | $SU(3)_c$ | $L$ | $B$ |
|---|---|---|---|---|---|---|
| $U = u_R^c$ | $1/2$ | $-\frac{2}{3}$ | $1$ | $\bar{3}$ | $0$ | $-\frac{1}{3}$ |
| $D = d_R^c$ | $1/2$ | $\frac{1}{3}$ | $1$ | $\bar{3}$ | $0$ | $-\frac{1}{3}$ |
| $Q = (u_L, d_L)$ | $1/2$ | $\frac{1}{6}$ | $2$ | $3$ | $0$ | $\frac{1}{3}$ |
| $L = (\nu_L, e_L)$ | $1/2$ | $-\frac{1}{2}$ | $2$ | $1$ | $1$ | $0$ |
| $E = e_R^c$ | $1/2$ | $1$ | $1$ | $1$ | $-1$ | $0$ |
| $H = (0, v + h/\sqrt{2})$ | $0$ | $\frac{1}{2}$ | $2$ | $1$ | $0$ | $0$ |

Table 9.1: *List of **Standard Model** Weyl fermions (quarks and leptons) and scalars, their charges under gauge interactions, and the lepton and baryon number accidental symmetries.*

neutrinos decouple while still relativistic, at the temperature

$$T_\gamma = T_\nu \sim v^{4/3}/M_{\rm Pl}^{1/3} \sim \text{few MeV}, \tag{9.1}$$

where $v$ is the electroweak VEV. At this temperature, the neutrino interaction rate, $\Gamma_\nu \sim \sigma_{\nu e} n_e \sim T^5/v^4$, becomes smaller than the Hubble rate $H \sim T^2/M_{\rm Pl}$. At neutrino decoupling the SM plasma consists mostly of $e^\pm$ and photons, so that the entropy density is proportional to $g_{*s} = 2 + 4(7/8) = 11/2$. Later, when the temperature drops to $T_\gamma \sim m_e$ the electrons annihilate with positrons, heating $\gamma$ but not $\nu$, while $g_{*s}$ decreases to $g_{*s} = 2$ (plasma is now composed only of $\gamma$). Since $Y_\nu = n_\nu/s$ stays constant, the neutrinos now have a lower temperature then photons, $T_\nu = T_\gamma(4/11)^{1/3}$. This means that the number densities are related, $n_\nu/n_\gamma = 3/11$ where $n_\nu$ is the number density for a single generation of neutrinos and antineutrinos. These cosmic background neutrinos have not yet been directly observed.

Neutrinos with mass $m_\nu \gg T_\nu$ contribute to the cosmological energy density as

$$\rho_\nu = \sum_\nu m_\nu n_\nu, \quad \text{or equivalently as} \quad \Omega_\nu h^2 = \frac{\sum_\nu m_\nu}{94 \text{ eV}}, \tag{9.2}$$

where for the latter equation we have used the standard definitions, see Appendix C. Here, the sum runs over the 3 generations. The cosmological DM density would then be reproduced if $\sum_\nu m_\nu \approx 11 \text{ eV}$.

However, the possibility of neutrinos being all of DM is excluded for several reasons. First of all, DM composed of neutrinos with $11 \text{ eV}$ mass would behave as warm dark matter, conflicting with the bounds in section 3.3.1. Actually neutrinos must be lighter, $m_\nu \lesssim \text{eV}$, as dictated by laboratory bounds on beta-decay spectra and (if neutrinos are Majorana) on neutrino-less double beta decay [438, 790]. This means that ordinary neutrinos cannot be all of the DM, and that neutrinos in cosmology must be a (hot) component with an abundance well below the DM abundance. Then bounds from matter clustering in cosmology and from the CMB imply $\sum_\nu m_\nu \lesssim \text{few } 0.1 \text{ eV}$, with the precise value depending on the assumed cosmological model and the employed dataset [791].

In summary, SM neutrinos cannot be DM because: 1) their weak interactions imply a nearly thermal abundance; 2) they are too light. Hypothetical extra '*sterile neutrinos*' can be heavier and don't have weak interactions: sterile neutrinos can be Dark Matter candidates, as will be discussed in section 9.2.2.

## 9.1.2 QCD bound states: quark matter?

Next, let's explore the possibility that exotic bound states of quarks could be the DM. Several authors, mostly around 1984, proposed that DM could consist of macroscopic objects, known as '**quark nuggets**'

(also called '**strangelets**' or '**nuclearites**'[1]) [103]. Quark nuggets would consist of a very large number of bound quarks, with approximately equal abundances of $u$, $d$, and $s$ quarks. They would be denser than normal nuclear matter, since the presence of three flavors of quarks, in contrast to the two quark flavors in normal matter, allows for a higher density that is still compatible with the Pauli exclusion principle. They would have a very large baryon number (above $10^2$ and possibly up to about $10^{57}$), a mass ranging from $10^{-8}$ kg to $10^{20}$ kg and a size from femtometers to meters. In a first approximation they would have zero electric charge, since the charges of the $u, d$, and $s$ quarks cancel each other out. However, because of the non-zero quark masses, the cancellation is not perfect. In particular, given their larger mass, the $s$ quarks are slightly less abundant hence strangelets typically have a net charge which is positive. This is welcome news, because negatively charged strangelets would have devastating consequences, eating up the nuclei they would encounter, while positively charged ones are electrostatically repelled by nuclei. The detailed computation of the net charge is quite involved and the result depends on a number of factors including the mass of the nugget. Since their intrinsic charge is non-zero, they are believed to be surrounded by a tightly-bound screening cloud of electrons, which makes them effectively neutral in most circumstances, and therefore good DM candidates.

These objects would have formed in the early Universe at the time of the QCD phase transition from regions of plasma between the bubbles of the confined phase, if such a phase transition were first order.[2] However, lattice simulations found that the QCD phase transition is instead a continuous crossover [104], so that quark matter cannot form in this most straightforward way. Di Clemente et al. (2024) [104] considered a speculative possibility that the quark matter might have formed from quark correlations at high temperatures, before chiral symmetry breaking. While this is motivated by the observations of nuclei formation in heavy ion collisions, dedicated calculations are still needed in order to establish whether the production of quark matter can really occur in this way already within the SM. Quark matter can form in extensions of the SM. For example, it could form during a first-order QCD-like phase transition that would occur if the electroweak symmetry breaking gets delayed and all the 6 quarks remain massless during the QCD phase transition (see Bai et al. in [104]). Another example are the axion extensions, which we discuss at the end of section 10.4.4.

A related issue is the one of stability. QCD simulations that include the effects of the finite size of the nuggets, of non-zero temperature and of non-zero masses for the constituent quarks, have not yet conclusively established whether or not a quark nugget would be stable, or if they instead slowly 'evaporate' and turn into ordinary nuclear matter. If the quark nuggets are indeed the more tightly bound state of matter, they could transform very dense objects made of normal nuclear matter, such as neutron stars, into strange matter. Although it is hard to find clear-cut ways of distinguishing strange stars from neutron stars, neutron stars do seem to behave as being made of neutrons, disfavoring the hypothesis of the existence of 'strange quark matter'.

The quark nuggets can be searched for in a number of ways. Notably, they could bind with nucleons and form very exotic and very heavy isotopes. In general, for large masses, they are subject to the constraints for macroscopic objects discussed in section 3.2 and figure 3.4. If they have a net electric charge, they are subject to the constraints discussed in section 3.3.2. In particular, they can be searched in cosmic rays, e.g., with the AMS detector. Their peculiarity is that they would have a very low charge-

---

[1]Sometimes these terms are used to distinguish objects of different sizes and masses within the category of 'quark matter' objects. However, since this distinction does not seem to be consistently used in the literature, we will treat the three terms as synonymous.

[2]In view of the strong QCD interactions, bubbles of the confined phase would have nucleated quickly, at a temperature slightly below the critical temperature, and released a large latent energy. In order not to overheat the Universe above the critical temperature, such bubbles would have expanded slowly. Since baryons are heavier than light quarks, the light quarks would have mostly remained in the deconfined region, getting compressed into small pockets of "quark matter" by the slowly expanding bubbles. The required first order phase transition would have happened in a world with either $N_f = 0$ or $3 \leq N_f \lesssim 3N_c = 9$ quarks that are lighter than the QCD scale. The real world has instead $N_f = 2$ light quarks, plus the intermediate-mass strange quark.

to-mass $Z/A$ ratio.

### 9.1.3   QCD bound states: hexa-quark DM?

A possibility closely related to the quark-nugget-DM discussed in the previous subsection is that DM is the so called 'hexa-quark' (or 'sexa-quark'), a neutral spin-0 bound state of six valence quarks, $S \sim uuddss$ [792]. This state is possibly relevant, because its assumed wave-function, being symmetric in the space components (and anti-symmetric in spin, color and flavour), could imply an unusually large binding energy and thus guarantee stability. Such a bound state has not yet been observed experimentally, but there is some limited evidence from lattice QCD calculations for a weakly-bound di-baryon state, $\Lambda\Lambda \sim (uds)(uds)$, with the same valence quark composition and binding energy of $\approx 5$ MeV (measured relative to $2m_\Lambda$), albeit computed at unphysical values of the quark masses.

A highly debated issue is the one of the effective coupling of $S$ to normal baryons such as neutrons, $y\,S\bar{n}\bar{n}$. For a typical QCD bound state one would expect $y \approx 4\pi$. However, several aspects of the phenomenology of $S$, discussed below, require a much smaller value for $y$ since this then implies cross sections $\sigma \sim y^2/\Lambda_{\rm QCD}^2$ many orders of magnitude smaller than the typical QCD cross sections. Some authors [792] argue that a very small value of $y$ is possible, as a consequence of the hypothetic small size of the bound state: if $S$ is small, its overlap with the the $nn$ state is suppressed, since the latter is 'large', as long as the short-range repulsive $nn$ nuclear interactions are approximated as hard-core (an approximation which has however been questioned). Other authors argue that such a behaviour, which would be unlike any other QCD bound state, is unrealistic, given that $S$ would have a mass typical of a QCD bound state, but a much smaller spatial size than typical of a QCD object.

From a phenomenological point of view, for $S$ to be a DM candidate and evade all bounds, one needs to consider:

▶ **DM stability**. The hexa-quark $S$ would be absolutely stable if $M_S < m_{\rm d} + m_e = 1.8761$ GeV (corresponding to a hexa-quark binding energy of $\approx 350$ MeV), such that even the $S \to de\bar{\nu}_e$ doubly-weak decay into deuteron $d$ is kinematically forbidden. For heavier $S$, in order for it to have a life-time longer than the age of the universe[3] one needs progressively smaller values of the coupling $y$ discussed above: $y \lesssim 10^{-4-5}$ up to $M_S < m_p + m_e + m_\Lambda \simeq 2.054$ GeV (binding energy $\approx 180$ MeV), above which single-weak decays $S \to p\Lambda e$ are kinematically open, and $y \lesssim 10^{-12}$ is needed.

▶ **Matter stability**. A hexa-quark $S$ lighter than about $1.85$ GeV is essentially excluded by the Super-Kamiokande bounds on $^{16}_{8}$O decays into $S$ plus other SM final particles. This decay would proceed too quickly through a double weak transition, unless the $nn \to S$ effective coupling is highly suppressed, $y \lesssim 10^{-10}$. The search for the $d \to Se^+\nu_e$ decay by the Sno experiment provides a slightly more stringent bound, which, however, does not reach $M_S > m_d - m_e = 1.8761$ GeV (where it would close the window between DM stability and matter stability) because $e^+$ needs at least $5-20$ MeV of energy to be above the Sno detection threshold. A related bound can be derived from the observed properties of the supernova explosion SN1987a. In the hot proto-neutron star, which forms during the supernova explosion, all the baryons would be processed into $S$ particles unless $y \lesssim 10^{-5-6}$, in the whole range of $S$ masses up to $M_S \approx 2200$ MeV.

▶ **Reproducing the cosmological DM abundance**. If $S$ has normal QCD-size couplings to baryons, the QCD interactions would initially keep $S$ in thermal equilibrium with other SM

---

[3]Note that the observational constraints on DM decays that result in $\gamma$ rays are much stronger, see fig. 6.15 in section 6.13.2. Decays of $S$ could result in hard photons, but the phenomenological implications have not been worked out in detail yet.

particles, $\Gamma/H \sim M_{\rm Pl}/m_p \sim 10^{20}$, and then decouple when $T$ is slightly below $M_S$. The observed cosmological DM abundance would then be reproduced via thermal freeze-out for a value of $M_S \approx 1.3\,{\rm GeV}$, which, however, is too low to be phenomenologically viable, as it is excluded experimentally from many sides. The cosmological DM abundance would instead be matched for $M_S \sim 1.8\,{\rm GeV}$ if the $S$ coupling to nucleons hypothetically were $y \sim 10^{-6}$, so that $S$ is never in thermal equilibrium [792]. Even assuming that such a highly suppressed effective coupling to baryons, $y \sim 10^{-6}$, is possible in QCD (see the discussion above), the $S$ would still be mostly excluded by the DM and matter stability constraints discussed above. Overall, only a very narrow range of masses, between about 1870 and 1880 MeV, remains.

▶ Compatibility with **direct detection bounds**. Unlike $S$ number-changing processes, which might possibly be suppressed, the $Sn \to Sn$ scattering cross section would unavoidably be of typical QCD size, $1/\Lambda^2_{\rm QCD}$. Hexa-quark DM would share the direct detection phenomenology of strongly interacting DM, discussed in section 5.1.2. In particular, it would not be constrained by underground direct detection searches, since it would not reach the detectors. It would, however, fall within the exclusion region of the X$_{\rm QC}$ experiment from searches in the upper atmosphere[4], unless the interactions reduce its velocity so that it is smaller than the usual cold DM velocity (in fact, the interaction cross section between $S$ and the gas in the Galaxy could be strong enough to drag dark matter in our local neighborhood into co-rotation with the solar system, reducing its effective speed). On the other hand, it may be boosted (see section 5.5.4) and be thus excluded by existing direct detection constraints (see Alvey, Bringmann & Kolesova (2022) [262]).

▶ **Production in decays.** For $M_S$ in the 1800 to 2050 MeV range, the bounds on too fast decays of hyper-nuclei that contain two $\Lambda$'s are especially important, and bound $y \lesssim 10^{-5}$. Furthermore, BaBar searched for $S$ in the process $\Upsilon \to S\bar{\Lambda}\bar{\Lambda}$, without success.

In conclusion, it remains to be seen whether QCD predicts both the large $S$ binding energy and, at the same time, the large suppressions of the $S$ couplings $y$ needed for $S$ to be a viable DM candidate. Future progress in lattice QCD calculations could help resolve this issue. Overall, a likely possibility is that $S$ is heavy enough to be unstable with QCD coupling $y \sim 4\pi$, in which case $S$ is just another typical QCD resonance, not relevant for DM.

## 9.1.4 EW bound states: top matter and dodeca-tops?

Next, let us move from QCD to the electroweak scale. In this case, a number of different quark bound states that could act as DM candidates have been explored in the literature.

One possibility are the so called **top bags**. The heaviest quark, the top quark $t$, gets its mass, $M_t = y_t v$, from a large Yukawa coupling to the Higgs boson, $y_t \approx 1$. Consequently, the Higgs boson can mediate a potentially significant force among top quarks. Had $y_t$ been large enough, the SM would have predicted a new composite state: 'top bags' containing $N_t \gg 1$ top quarks. Inside the top bags the Higgs boson would have smaller vacuum expectation value $v_{\rm in}$ than in the usual low temperature vacuum, $v_{\rm in} < v$ (possibly even $v_{\rm in} = 0$). The top quarks would therefore be lighter in the region inside the top bags, $M_t^{\rm in} < M_t$. Omitting $\mathcal{O}(1)$ factors, the energy of a top bag with radius $R$ that contains $N_t$ non-relativistic tops can be estimated as their kinetic energy plus the vacuum energy,[5] giving for the binding energy $E_B$

$$E_B \sim -N_t^{5/3}/(M_t^{\rm in} R^2) - \lambda(v^2 - v_{\rm in}^2)^2 R^3 + N_t(M_t - M_t^{\rm in}), \tag{9.3}$$

---

[4] See however Xu and Farrar (2021) in [792], which claim significantly relaxed bounds when allowing for resonant DM/nucleus scattering, and including finite size effects for nuclei.

[5] These quantities will be more precisely discussed in section 10.5.1 and 10.5.2, where composite bound states more general than 'top bags' are studied, including their possible formation mechanisms in cosmology.

where $\lambda$ is the quartic Higgs self-coupling. We defined $E_B$ as the difference with respect to free top quarks, such that $E_B > 0$ would kinematically forbid the decay of a top bag into free tops. A very large binding energy $E_B \simeq N_t M_t$ would correspond to an almost massless top bag, that may be stable, if its mass $M = N_t M_t - E_B > (N_t/3)m_p$ (since baryon number is conserved, while the top bag has baryon number equal to $N_t/3$). If charge is neutralized by electrons, stable top bags might be a DM candidate. Maximising $E_B$ with respect to $R$ and allowing the maximal $N_t \sim (M_t R)^3$ compatible with non-relativistic tops (otherwise the Higgs force becomes ineffective) shows that $E_B > 0$ needs $y_t \lambda^{1/4} \gtrsim 1$. Including order unity factors, this condition is $M_t \gtrsim \mathrm{TeV}$ for $M_h \simeq 125\,\mathrm{GeV}$ [789], while $M_t \approx 172\,\mathrm{GeV}$ in the SM. Therefore, top bags do not exist for the SM values of $M_t$ and $M_h$.

Froggatt and Nielsen [789] considered the possibility that the Higgs force might be strong enough to form a bound state made out of 6 $t$ and 6 $\bar{t}$ quarks, a **dodeca-top**, since 6 particles allow to fill an optimal $s$-wave state (3 colours times 2 spins). Furthermore, they speculated that its binding energy $E_B$ might be larger than $12M_t$, making dodeca-top tachyonic, thus forming a condensate, $\langle(t\bar{t})^6\rangle \neq 0$ that locally modifies the Higgs vacuum expectation value $v$. In this scenario the DM candidate are balls of false vacua that contain many ordinary atoms but now with smaller masses due to a smaller $v$, which makes them stable [789]. Approximating a dodeca-top with 6 $t\bar{t}$ pairs leads Froggatt and Nielsen (2003) to rather crudely estimate that a top Yukawa coupling larger than its observed value is required, $y_t \gtrsim [0.13(4\pi)]^{1/2} \approx 1.3$. This would imply that no such object exists in the SM.

Some patterns of symmetry breaking imply magnetic monopoles stable for topological reasons and formed during cosmological phase transitions. However the SM electro-weak phase transition $\mathrm{SU}(2)_L \otimes \mathrm{U}(1)_Y \to \mathrm{U}(1)_{\mathrm{em}}$ does not lead to magnetic monopoles because (in the language of section 9.6, where monopoles will be discussed) the associated second homotopy group is trivial, $\pi_2(\mathrm{SU}(2)) = 0$. Furthermore, the transition happens cosmologically as a cross-over (see e.g. [793]).

Nevertheless, theorists explored the possibility that the SM might predict a variety of more complicated solitonic objects involving the Higgs and the $W, Z$ bosons. Despite early claims (Cho and Maison (1996)), the conclusion seems to be that in the SM no stable solitons with finite energy exist [789]. Extra effective operators or other new physics are needed in order to arrive at stable electroweak solitons, with mass $M$ controlled by the new-physics scale [789]. One interesting example is gauge unification: $\mathrm{SU}(5) \to G_{\mathrm{SM}}$ predicts magnetic monopoles with mass $M \sim 10^{16}\,\mathrm{GeV}$. Such magnetic monopoles cannot be DM candidates, because they would be accelerated by galactic magnetic fields, erasing them [794]. The persistence of the Milky Way magnetic fields implies that $\mathrm{SU}(5)$ monopoles have a density 6 order of magnitude below the galactic DM density in eq. (2.11) [794]. Direct experimental searches set a mildly stronger bound [794].

One more possibility within the SM is that an extra Higgs vacuum might exist around the Planck scale, with energy density that is degenerate with the electroweak vacuum. However, even if this is the case, this would still not give rise to $Q$-balls or other stable bound states, as the Higgs boson is much lighter in the physical electroweak vacuum.

## 9.1.5   Gravitational/nuclear bound states: neutron balls?

Neutrons are electrically neutral, massive unstable baryons. Free neutrons cannot be the DM because they decay, and because the baryon abundance is measured to be below the DM abundance (see section 1.3.5). Neutrons become stable when packed into dense nuclear matter with a large enough binding energy. We here explore the possibility that such objects could be a DM candidate. In order to have an idea of possible concrete realizations, it is worth exploring briefly the general conditions under which nuclear matter forms.

Nuclear matter with density $\rho \sim m_p^4$, and containing $Z$ protons and $A - Z$ neutrons, only forms in two regimes, *i)* for $A - Z \approx Z \lesssim \sim 100$, and *ii)* for $A \gtrsim (M_{\mathrm{Pl}}/m_n)^3 \sim 10^{57}$, $Z \approx 0$:

○ The first nuclear matter regime, $A - Z \approx Z \lesssim 100$, consists of regular nuclei. Nuclei exist along the so called 'valley of stability' in the $A, Z$ parameter space, i.e., when $Z \approx A/2$. This is because only the $np$ nuclear forces can be attractive, while the $nn$ and the $pp$ nuclear forces are repulsive. Since protons are charged, the long-range Coulomb repulsion dominates at large $Z$ over the QCD binding energy per nucleon $\sim m_{u,d}$, and forbids stable nuclei with $Z \gtrsim (m_{u,d}/\alpha\Lambda_{\mathrm{QCD}})^{3/2} \sim 100$ protons. This is why in the overview figure, fig. 3.1, stable nuclei exist only at the very beginning of the red dashed line, i.e, for the smallest masses of the objects, just above the quantum boundary. Such charged nuclei cannot be DM.

○ The second nuclear matter regime consists of objects with radii $R$, containing $Z = 0$ protons, and a number $A$ of neutrons that is large enough for attractive gravitational force to win over the $nn$ nuclear repulsion. This results in a per nucleon binding energy that is large enough to kinematically block $n \to pe\bar{\nu}$ decays, $U_B/A \gtrsim \Delta = m_n - m_p - m_e$. The gravitational binding energy is given by

$$U_B \approx \frac{3}{5}\frac{G\mathcal{M}^2}{R}, \qquad \text{where} \qquad \mathcal{M} = Am_n \qquad \text{and} \qquad R \approx d\,A^{1/3}. \qquad (9.4)$$

Since the distance $d$ among neutrons must be $d \gtrsim 1/m_\pi$ in order to avoid nuclear repulsion, gravity is strong enough to block neutron decays only for $A \gtrsim 2\,(M_{\mathrm{Pl}}/m_n)^3(\Delta/m_\pi)^{3/2}$, i.e., $\mathcal{M} \gtrsim 10^{-3}M_\odot$ and $R \gtrsim M_{\mathrm{Pl}}\Delta^{1/2}/m_\pi^{3/2}m_n \approx 5$ km. Hence, **small gravitational neutron balls** with a size of a few km and a mass of a thousandth of the Sun could in principle be stable DM candidates.[6] However, even if they could somehow form before BBN, the micro-lensing bounds in fig. 3.3 (see section 3.1) exclude that they can be all of the DM. It seems unlikely that the clustering of neutron balls allows to avoid this exclusion, since the bounds are stringent and extend well below and above the predicted mass $\mathcal{M}$.

## 9.1.6 Gravitational bound states: black hole DM in the SM?

The discussion in the previous subsections leaves us with only one remaining DM candidate in the SM: primordial black holes (PBH). These can potentially be made out of just the ordinary SM baryons, although in general their production in the early Universe does depend on non-SM inflationary physics.

The range of masses for which PBHs are an allowed DM candidate, as well as all the relevant constraints, have been discussed in section 3.1.1. The general mechanism which results in a PBH formation was presented in section 4.6: it requires large primordial density fluctuations of ordinary matter on small scales, which then gravitationally collapse and form black holes.

Within the SM, if one extrapolates the effective Higgs potential $V_{\mathrm{SM}}(h) \approx \lambda(h)h^4/4$ to ultra-large field values, including loop corrections, this reaches a maximum around $h \sim 10^9\,$GeV and then starts decreasing, indicating the existence of an extra vacuum at $h \sim M_{\mathrm{Pl}}$ (in addition to our electroweak vacuum with $h \ll M_{\mathrm{Pl}}$). Espinosa et al. (2017) in [177] proposed that this instability might result in large primordial inhomogeneities, which are needed in order to form primordial black holes. For this to work, one needs to *assume* that the Higgs, at $N \sim 20$ $e$-folds before the end of inflation, has a *highly homogeneous* vacuum expectation value $h_{\mathrm{in}}$ that is slightly larger than the value at which the potential barrier in the Higgs potential is maximal. The classical evolution of the Higgs field, $\ddot{h} + 3H_{\mathrm{infl}}\dot{h} + V'_{\mathrm{SM}}(h)$, then starts to dominate over quantum fluctuations, $V'_{\mathrm{SM}}(h_{\mathrm{in}}) \gtrsim H^3_{\mathrm{infl}}$, and the Higgs rolls down the potential toward larger field values. During this phase, in regions with larger $h$ the Higgs rolls faster towards the vacuum at $h \sim M_{\mathrm{Pl}}$. This means that, assuming the Bunch-Davies vacuum when the

---

[6]Note that standard astrophysics only leads to the formation of neutron stars above the Chandrasekhar bound, $A \gtrsim (M_{\mathrm{Pl}}/m_n)^3$ or, more precisely, $\mathcal{M} > 1.4M_\odot$. In fig. 3.1 this is around the black hole end of the red dashed line.

fall starts, small quantum inhomogeneities $\delta h_k$ at wave-number $k$ get amplified. Indeed, the evolution equation for the inhomogeneities in the Higgs field, $\ddot{\delta h}_k + 3H_{\text{infl}}\dot{\delta h}_k + k^2\delta h_k/a^2 + V''_{\text{SM}}(h)\delta h_k = 0$, implies that $\delta h_k$ grows as $\dot{h}$ on super-horizon scales, as can be seen by operating a time derivative on the classical equation for $h$. Matching approximate solutions at horizon crossing $t_k$ one finds for inhomogeneities at a later time $t$ to be given by $\sqrt{2k^3}|\delta h_k(t)| \approx H_{\text{infl}}\dot{h}(t)/\dot{h}(t_k)$.

When inflation ends, reheating adds a large thermal mass to the effective Higgs potential. Under certain conditions this brings the Higgs back from $h \sim M_{\text{Pl}}$ to the physical SM vacuum at the origin, $h = 0$. Given the shape of the SM Higgs potential, the roll-down phase can last about 20 $e$-folds, leading to PBH with mass $M \sim e^{2N}M_{\text{Pl}}^2/H_{\text{infl}}$ in the range where PBH can constitute all of the DM. This process generates the desired amount of PBHs if the primordial fluctuations in the curvature $\zeta$ are enhanced by $\sim 10^3$. To achieve this, $h_{\text{in}}$ must be tuned at the $\sim 10^{-3}$ level, such that the roll of the Higgs towards the Planckian minimum is stopped just before it would reach its maximal value from which it can still be driven back into the SM vacuum by thermal effects.

However, while inflation can produce a roughly constant $h_{\text{in}}$, it also produces inflationary fluctuations $\delta h_{\text{in}} \sim H_{\text{infl}}/2\pi$. Since $H_{\text{infl}} \sim h_{\text{in}}$ is needed, these fluctuations are too large and prevent the needed accurate homogeneous tuning of $h_{\text{in}}$. If one of the $\sim e^{3(60-20)}$ disconnected Hubble patches falls into $h \sim M_{\text{Pl}}$, it drives the whole universe in the wrong vacuum (see Gross et al. in [177]). Some extra mechanism beyond the SM is needed to avoid fluctuations in $h_{\text{in}}$, such as an extra ultra-heavy scalar suitably coupled to the Higgs [177]. Beyond the SM, different inflation mechanisms can provide large inhomogeneities and PBH as DM, as discussed in section 4.6.1.

Alternatively, the SM Higgs potential $V_{\text{SM}}(h)$ could feature an inflection point for a value of $h$ close to the Planck scale. An inflection point arises for tuned values of the SM parameters (mostly $M_h, M_t$ and $\alpha_s$) such that the Higgs quartic coupling $\lambda(h)$ RG evolves to an almost vanishing value at a particular high scale [178]. In the SM, the RG scale at which $\lambda(h)$ is minimal (and thereby the inflection point is reached) is $h \sim 0.1\bar{M}_{\text{Pl}}$. This Higgs field value and the corresponding potential energy $V_{\text{SM}}(h)$ are both too large for the Higgs to drive the observed inflation, and also for the Higgs inflation to generate PBHs that could be the DM [178].

## 9.2  Dark Matter not charged under the SM group

### 9.2.1  Scalar singlet DM

The most economical DM model, at least in terms of new degrees of freedom, consists of adding a real neutral singlet scalar $S$ to the SM [795], and imposing an ad-hoc global symmetry $S \to -S$, to make $S$ stable. The most general renormalizable Lagrangian is then given by

$$\mathscr{L} = \mathscr{L}_{\text{SM}} + \frac{(\partial_\mu S)^2}{2} - \frac{m_S^2}{2}S^2 - \lambda_{HS}S^2|H|^2 - \frac{\lambda_S}{4}S^4 \, , \tag{9.5}$$

and contains, besides the kinetic and mass terms for $S$, also a quartic scalar interaction between $S$ and the Higgs doublet $H$, and the phenomenologically mostly irrelevant quartic self-coupling, $S^4$.[7] After the higgs field obtains its vev, $v \approx 174$ GeV, the DM mass is given by $M^2 = m_S^2 + 2\lambda_{HS}v^2$. This must be positive, so that $\langle S \rangle = 0$, and $S$ remains stable.

---

[7]Global symmetries might be broken by Planck mass suppressed operators, in which case DM could be made stable by considering a more complicated model, based on a local $\mathbb{Z}_2$ symmetry. Such a $\mathbb{Z}_2$ symmetry could be, for instance, a remnant of a dark U(1) gauge group, under which $S$ has a charge 1, spontaneously broken by the vacuum expectation value of another scalar, $S'$, with charge 2. This set-up would result in a distinct phenomenology: the complex $S$ would contain two mass–split components, with the splitting induced by the $SSS'^*$ cubic interaction (see Baek et al. (2014) in [795]).

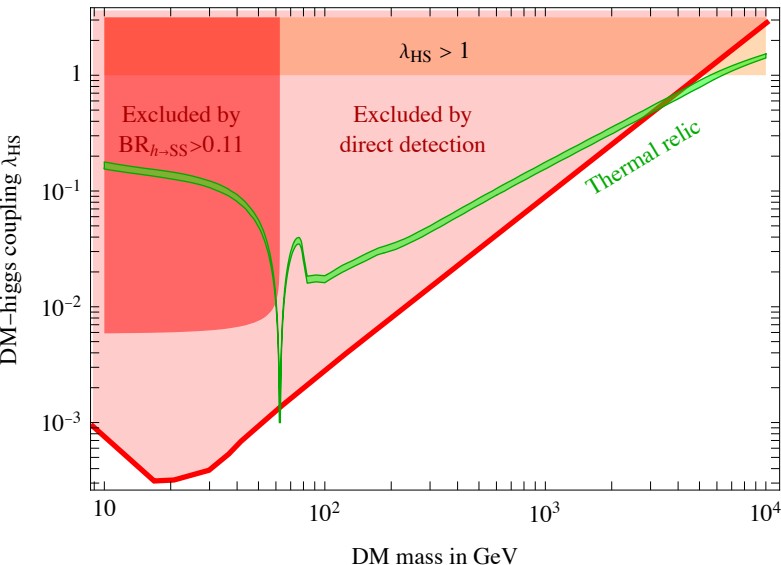

Figure 9.2: **Scalar singlet DM**. *In the* $(M, \lambda_{HS})$ *plane we plot the region excluded by direct detection, the band favoured by the thermal DM abundance, and the region where the quartic coupling is bigger than unity.*

The spin-independent direct detection DM scattering cross section is given by [795]

$$\sigma_{\rm SI} = \frac{\lambda_{HS}^2 m_N^4 f_N^2}{\pi M^2 M_h^4}, \tag{9.6}$$

where $f_N \approx 0.3$ is the nucleon matrix element (see section 5.1.6), $M_h \approx 125$ GeV is the higgs boson mass, $m_N = 0.939$ GeV is the nucleon mass and for simplicity $m_N \ll M$ was assumed in order to shorten the expression. The DM relic abundance is set by the thermal freeze-out (see section 4.1), where the non-relativistic $s$-wave $SS$ annihilation into SM particles can proceed in two ways: (i) via $SS \to h \to f\bar{f}$, where $f$ is any kinematically accessible SM final state, i.e., any SM fermion or gauge boson for which $m_f < M$, and (ii) via direct $SS \to hh$ annihilation, possible if $M_h < M$. The result is [795]

$$\sigma v = \frac{16\lambda_{HS}^2 v^2}{\left(4M^2 - M_h^2\right)^2 + M_h^2\left(\Gamma_h(M_h) + \Gamma_{h \to SS}\right)^2} \frac{\Gamma_h(2M)}{2M} + \frac{\lambda_{HS}^2}{64\pi M^2}\sqrt{1 - \frac{M_h^2}{M^2}}\Theta(M - M_h), \tag{9.7}$$

where the $h \to SS$ partial decay width, for $M_h > M$, is given by

$$\Gamma_{h \to SS} = \frac{\lambda_{HS}^2 v^2}{4\pi M_h}\sqrt{1 - 4\frac{M^2}{M_h^2}}. \tag{9.8}$$

The second term in eq. (9.7) accounts for the $SS \to hh$ annihilation channel, with $\Theta$ the step function. The first term in (9.7) accounts for the $SS \to h \to f\bar{f}$ channel, and has the Breit-Wigner form due to the intermediate Higgs resonance. It is expressed in terms of $\Gamma_h(m)$, the total decay width of a SM Higgs boson, had it had mass equal to $m$. The tabulated values for $\Gamma_h(m)$ can be found, e.g., in [796]. For $M \gg M_W$ the expression for the annihilation cross section eq. (9.7) tends to its $SU(2)_L$-invariant limit,

thanks to the apparent non-decoupling of the $\Gamma_h(m)$ factor. Note that the $S^4$ quartic coupling helps in maintaining the kinetic equilibrium also close to the Higgs resonance, $M \approx M_h/2$, where the annihilation is resonantly enhanced, such that this regions remains allowed by direct detection bounds.

In fig. 9.2 we show with dark red shading the region in the $(M, \lambda_{\text{HS}})$ plane that is excluded by the bounds on the invisible width of the higgs, i.e., in the excluded region $\text{BR}(h \to SS) > 0.11$ [694]. The region excluded by direct DM searches is shown with light red shading, and the band where the thermal relic DM density reproduces the measured DM density $\Omega_{\text{DM}}$ by a green band. Indirect detection bounds could also be added on the same plane but turn out not to be very relevant.

In this simple model a single coupling, $\lambda_{HS}$, controls both the relic density abundance and the direct detection scattering cross section, since both involve either a Higgs exchange or Higgs production. It is therefore possible to fully test the model in different, complementary ways. Assuming there are no other production mechanism beside thermal DM production, the model is allowed only for DM that is heavy enough, $M \gtrsim 3\,\text{TeV}$. There is also an upper bound on DM of about 7 TeV, since for larger DM masses the thermal annihilation cross section requires large couplings, $\lambda_{HS} \gtrsim 1$, and thus runs afoul of perturbativity bounds. Since only a finite DM mass interval remains viable, future experiments can in principle cover the whole parameter space of the model.

In the complex $S$ version of the model, DM can be stable after imposing either $\mathbb{Z}_2$, $\mathbb{Z}_3$, or U(1) global symmetries, or an invariance under complex conjugation $S \to S^*$ [797]. The latter symmetry remains unbroken even for $\langle S \rangle \neq 0$, such that $\text{Im}\,S$ remains a stable DM candidate. The Lagrangian for the $\mathbb{Z}_3$ model is given by

$$\mathscr{L} = \mathscr{L}_{\text{SM}} + |\partial_\mu S|^2 - m_S^2 |S|^2 - \lambda_{HS} |S|^2 |H|^2 - \lambda_S |S|^4 + A(S^3 + S^{*3}). \tag{9.9}$$

DM is stable despite the cubic $S^3$ interaction, as the action is invariant under a $S \to e^{2\pi/3} S$ realising a $\mathbb{Z}_3$ symmetry. If DM is the pseudo-Goldstone phase of a complex $S$ with spontaneously broken U(1) symmetry, its interactions get suppressed at low energy, reducing direct detection bounds [798].

## 9.2.2 Fermionic singlet DM: 'sterile neutrino'

Another minimal choice for a DM model is to add to the SM an extra neutral Weyl fermion $N$. Here, one faces a choice of whether or not to impose the ad-hoc $N \to -N$ symmetry, similarly to what was done in section 9.2.1 for the scalar singlet DM. Imposing the $N \to -N$ symmetry makes $N$ stable, but also means that there are no renormalizable interactions between $N$ and the SM (in contrast to the scalar singlet DM, where one can still write down the Higgs quartic $S^2|H|^2$ term, see eq. (9.5)), making the production of DM in the early universe an unresolved question. One could introduce both $N$ and an extra scalar singlet $S$ (or, alternatively, a vector singlet); however the resulting models [799] would lead us away from minimality.

Dropping the requirement of an $N \to -N$ symmetry is a much more widely discussed possibility, that goes under the name of a *sterile neutrino* or *right-handed neutrino*, and we thus focus on it. The attractive feature is that one can now write down a Yukawa interaction between $N$, the left-handed leptons $L$ of the SM, and the Higgs doublet $H$:

$$\mathscr{L} = \mathscr{L}_{\text{SM}} + \bar{N} i \slashed{\partial} N + \left( M \frac{N^2}{2} + y\,NLH + \text{h.c.} \right). \tag{9.10}$$

In fact, adding two or three right-handed neutrinos $N_i$, with a Majorana mass matrix $M$ and Yukawa couplings $y_{ij}\,N_i L_j H$ is the simplest extension of the SM that at low energies reproduces the small sizes of neutrino masses via the see-saw mechanism, $m_\nu = v^2\,y^T M^{-1} y$, after the heavy degrees freedom $N_i$ are integrated out. In this context, one often considers heavy sterile neutrinos with masses $M \sim 10^{10}\,\text{GeV}$

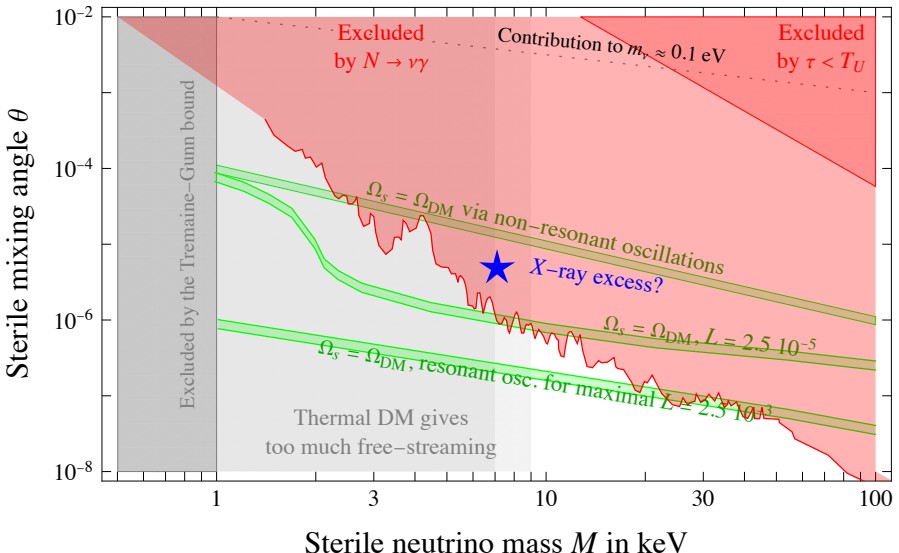

Figure 9.3: **Sterile neutrino DM**. *The red region is excluded from DM decays into X-rays [107], while in the dark red region DM also has a lifetime that is shorter than the age of the Universe. Production via non-resonant neutrino oscillations reproduces the DM abundance in the upper green band, while resonant oscillations give the lower green bands, which assume the indicated lepton asymmetry [107]. The light gray region below $M < 9$ keV is excluded by free-streaming, if DM has thermal velocity; such bound becomes somewhat weaker, if DM has a mildly sub-thermal velocity distribution (mid-dark gray). Below the dotted line the $N$ contribution to the active neutrino masses is below their observed values. The star indicates the possible $3.5$ keV excess, discussed in section 8.2.3.*

such that $y \sim 1$. This set-up also leads to a successful baryogenesis via leptogenesis already in its minimal implementation (see [164] for reviews).

For DM the more interesting region of the parameter space is if the sterile neutrino has a mass $M \gtrsim$ keV [107], just above the Gunn-Tremaine bound for fermionic DM, see eq. (3.8). This then leads to a viable DM candidate, albeit a decaying one. For such light $N$ the Yukawa coupling $y$ needed for the see-saw becomes small enough so that the sterile neutrino is stable on cosmological time-scales. Working to first nonzero order in small $y$, the interactions of $N$ with the light SM fields are the same as for the active SM neutrinos, but suppressed by the mixing angle between the sterile neutrino and the active neutrinos, $\theta = yv/M \ll 1$.[8] For $M < m_e$ the dominant decay mode of $N$ is into active neutrinos, with the decay width given by [800]

$$\Gamma(N \to 3\nu) = \frac{G_F^2 M^5}{96\pi^3}\theta^2 \approx \frac{\theta^2}{33\,T_U}\left(\frac{M}{\text{keV}}\right)^5. \tag{9.11}$$

This partial decay width (that dominates the DM lifetime $\tau$) must be longer than the age of the universe, $\tau \approx 1/\Gamma(N \to 3\nu) \gtrsim T_U = 13.7$ Gyr. As shown in fig. 9.3, this bound would allow for sterile neutrino DM that gives the right contribution to neutrino masses (which occurs along the dotted black line in the upper part of the figure). However, significantly stronger bounds on sterile neutrino decays follow from the modes that include photons, since photons are much easier to detect. The dominant such decay

---

[8]We use a single angle $\theta$ that describes mixing of $N$ with a particular linear combination of $\nu_e, \nu_\mu, \nu_\tau$, since for our discussion the SM neutrino flavors are irrelevant. For production of $N$, for instance in $\nu_e$ or $\nu_\mu$ scattering on target, on the other hand, one needs to distinguish between different mixing angles for each SM neutrino flavor.

mode is $N \to \nu\gamma$, arising at one-loop level. The corresponding partial decay width is

$$\Gamma(N \to \nu\gamma) = \frac{9\alpha G_F^2 M^5}{256\pi^4}\theta^2 \approx \frac{\theta^2}{4250\,T_U}\left(\frac{M}{\text{keV}}\right)^5. \tag{9.12}$$

Observations require $\tau\,\text{BR}(N \to \nu\gamma) = 1/\Gamma(N \to \nu\gamma) \gtrsim 10^8\,T_U$, with the precise bound shown in fig. 9.3 as a light red shaded region. This implies that the $N$ contribution to neutrino masses, $m_\nu = M\theta^2$, is smaller than about $0.04\,\text{eV}(\text{keV}/M)^4$, and thus smaller than the largest difference between SM neutrino masses. This means that there needs to be additional contributions to SM neutrino masses beyond the see-saw type contribution due to mixings with the sterile neutrino DM.

## Freeze-in via non-resonant oscillations

The same mixing angle $\theta$ also controls the production of sterile neutrino DM in the early Universe. In the dense plasma, active-sterile oscillations are averaged by the decoherent scatterings on matter, and the production process can be seen as a version of the freeze-in mechanism (section 4.2.2). At a temperature $T \ll M_Z$ the interaction rate of the sterile neutrino with the thermal bath is given by $\Gamma_s \approx \theta_m^2 \Gamma_\nu$, where $\Gamma_\nu$ is the interaction rate of any of the three active neutrinos [107]. The modified active/sterile mixing angle[9], $\theta_m \approx \theta/(1 + \mathcal{O}(1)T^6 G_F^2/\alpha M^2)$, includes the effects due to finite density and finite temperature. The ratio between $\Gamma_s$ and the Hubble rate,

$$\frac{\Gamma_s}{H} \approx \theta_m^2\left(\frac{T}{3\,\text{MeV}}\right)^3, \tag{9.13}$$

is maximal at $T_* \approx 130\,\text{MeV}(M/\text{keV})^{1/3}$ and results in the following amount of sterile neutrinos, as measured in terms of new fermionic, relativistic degrees of freedom $\Delta N_\nu$ (contributions to $\Delta N_\text{eff}$, see eq. (C.27) in App. C.3):

$$\Delta N_\nu \approx \min\left(1, \max_T \frac{\Gamma_s}{H}\right) \approx \min\left(1, 10^6\,\theta^2\,\frac{M}{\text{keV}}\right). \tag{9.14}$$

The mass abundance at low temperatures, when sterile neutrinos are well in the non-relativistic regime, is then $\Omega_s h^2 = \Delta N_\nu \cdot M/94\,\text{eV}$, see eq. (9.2). This production mechanism is also referred to as the **Dodelson-Widrow mechanism**. It leads to sterile neutrinos with a roughly thermal velocity.

The upper green band in fig. 9.3 shows the region of parameter space in which sterile neutrino production via non-resonant oscillation reproduces the observed cosmological DM abundance. However, a combination of the bound from $X$-rays due to DM decays (red region) and on DM free-streaming (eq. (3.9), gray region) excludes the possibility of a DM production via non-resonant oscillations. The following assumptions went into the prediction for DM abundance: *i)* that there were basically no sterile neutrinos in thermal plasma at temperatures above about 1 GeV, *ii)* that the only sterile neutrino interactions are the ones induced from mixing with the active SM neutrinos; *iii)* that the Universe has no lepton asymmetry at temperatures below 1 GeV. Violating one or more of the above assumptions can change the predictions, where a notable example is the freeze-in via resonant oscillations, discussed below.

## Freeze-in via resonant oscillations

Since the sterile neutrino is heavier than the active neutrinos, resonant oscillations can only happen due to in medium effects. If the thermal plasma has a lepton asymmetry, this creates an effective mass term

---

[9]The full expressions provided in the literature [107] reduce to this compact approximated formula for $\theta \ll 1$.

for neutrinos.[10] The changed dispersion relation for active neutrinos can be similar to the dispersion for sterile neutrinos, for some particular momenta. This level crossing transfers the leptonic excess from active neutrinos into the sterile neutrinos. The resulting sterile neutrino DM relic abundance is roughly proportional to the lepton asymmetry, $L = (n_\nu - n_{\bar\nu})/(n_\nu + n_{\bar\nu})$ (we assume that there is no lepton asymmetry in the charged leptons sector, in agreement with observations). This mechanisms is also known as the **Shi-Fuller mechanism**. For it to work, $L$ needs to be much larger than the baryon asymmetry, by at least 5 orders of magnitude.

Furthermore, the minimal model with 3 sterile neutrinos can also produce baryogenesis together with the large lepton asymmetry $L$, provided that the CP-asymmetry is enhanced by having 2 quasi-degenerate heavier sterile neutrinos. The scatterings that produce the two heavy sterile neutrinos first result in a lepton asymmetry, which then gets converted into a baryon asymmetry via sphalerons. This process is active only at temperatures above the electroweak scale, i.e., before the sphalerons decouple [107]. The scatterings at lower temperatures still continue to produce a lepton asymmetry, which thereby is bigger than the baryon asymmetry — possibly almost as big as the maximal lepton asymmetry allowed by the BBN constraints, $L \lesssim 2.5 \, 10^{-3}$ [107]. This lepton asymmetry then gets converted into relic abundance of the lightest sterile neutrino state, $N_1$, via resonant oscillations.

If the sterile neutrino DM state $N_1$ is produced through resonant oscillations, it tends to have a sub-thermal average velocity, since the level crossing happens predominantly for lower $N_1$ momenta. The sterile neutrino DM is then less warm than if it were produced via a thermal freeze-out, and thereby the bounds from free-streaming are weaker, roughly $M \gtrsim 7 \, \mathrm{keV}$ (Baur et al. (2017) in [107]).

### Freeze-in via decays

The complications, such as the need for a large lepton asymmetry $L$, which are encountered in the minimal scenario, motivate the investigations of other, non-minimal, production mechanisms that could result in a production of relatively cold sterile neutrinos [107]. The simplest possibility would be a freeze-in production from the Higgs decays. However, in this case eq. (4.46) implies a too large value of the Yukawa coupling $y$ in eq. (9.10), and thereby a too fast $N \to \nu\gamma$ decay according to eq. (9.12). One needs to introduce extra particles, such as a scalar singlet $\phi$ with a Yukawa interaction $\phi N^2$ or a vector lepto-quark $\mathcal{U}$ with a gauge-like interaction $(\bar{u}_R^i \gamma^\mu N)\mathcal{U}_\mu$, which then set the relic abundance. Note that, similarly to the freeze-in via resonant oscillations, the freeze-in production of $N$ via decays, with a partially closed phase space, also gives rise to the sterile neutrino DM with a sub-thermal velocity distribution. The bound on the mass of warm DM quoted in eq. (3.9) can therefore be appreciably relaxed, so that the sterile neutrinos could even have a mass $M = 7 \, \mathrm{keV}$. This is, for instance, needed to fit the possibly anomalous 3.5 keV $X$-ray line discussed in section 8.2.3.

The viable parameter space of models can be further enlarged, if one can invoke some mechanism that dilutes the abundance of sterile neutrinos after the freeze-in era completes, such as an entropy release into other particles at some later time [107, 147].

## 9.2.3 Vector singlet DM

Building on the theme of DM models with minimal number of extra degrees of freedom discussed in the previous sections, one can also consider adding to the SM a vector singlet $A$ [802]. A $A^\mu \to -A^\mu$ symmetry (either imposed by hand or inherited by other reasons, see section 10.6 and [802]) can guarantee

---

[10]More precisely, thermal effects induce a term in a dispersion relation that is linear in the neutrino energy, and to which the forward scatterings of neutrinos on neutrinos in a thermal plasma give opposite contributions then do the scatterings of neutrinos on antineutrinos. This effect therefore vanishes for $n_\nu = n_{\bar\nu}$ [801].

the stability of $A$. Then, a possible effective Lagrangian is

$$\mathcal{L} = \mathcal{L}_{\mathrm{SM}} - \frac{1}{4}F^{\mu\nu}F_{\mu\nu} + \frac{M_A^2}{2}A_\mu A^\mu + \frac{\lambda_{HV}}{2}A_\mu A^\mu |H|^2 - \frac{\lambda_V}{4}(A_\mu A^\mu)^2 \;, \qquad (9.15)$$

up to issues related to gauge invariance. The mass term $M_A$ can be generated by a Stückelberg or Higgs mechanism in the hidden sector. Extra scalars are needed in the latter case.

   This model features a phenomenology similar to the scalar singlet DM discussed in section 9.2.1 [802]. The typically considered mass range is around the weak scale. In the sub-GeV mass range, the model corresponds to the dark photon, see section 9.4.1.

## 9.3   Dark Matter charged under the SM group

### 9.3.1   Charged Dark Matter

Observational constraints essentially exclude DM with an electric charge comparable to the electron charge $e$. DM could, however, carry a fraction of a unit charge. This possibility goes under the name of milli-charge DM, and was reviewed in section 3.3.2.

### 9.3.2   Colored Dark Matter

Moving to color, could additional particles $Q$ charged under QCD constitute the DM? Any such colored particles would be confined inside hadrons, i.e., bound states that are singlets under $\mathrm{SU}(3)_c$. Given that hadrons continue to have strong interactions, there are stringent bounds on such scenarios, even in the least constrained case where additional colored particles only form neutral hadrons so that the bounds on charged DM are avoided. The hadrons containing valence light quarks $q = \{u, d\}$ (such as $QQq$ and $Qqq$) and/or gluons are $\approx 1/\Lambda_{\mathrm{QCD}}$ in size. The cross section for scattering of such DM on nucleons is therefore $\sigma_{\mathrm{SI}} \approx \pi/\Lambda_{\mathrm{QCD}}^2 \approx 1.6\ 10^{-26}\ \mathrm{cm}^2$. Such DM candidates have been excluded up to masses $M \sim 10^{15}\ \mathrm{GeV}$ or slightly higher, as discussed in section 5.1.2.

   The remaining allowed DM candidates are neutral hadrons that have as constituents *only* the new stable colored particles $Q$, with masses $M_Q \gg \Lambda_{\mathrm{QCD}}$. The hadrons are then made out of, e.g., $QQ$ if $Q$ is a color octet, and out of $QQQ$ if $Q$ is a triplet, with both options resulting in viable DM candidates [803]. The hadrons are bound states of a Coulomb-like potential, with large binding energies, $E_B \approx \alpha_{\mathrm{s}}^2 M_Q$, and of small sizes on the order of $1/\alpha_{\mathrm{s}}M_Q$, such that the bound states have small residual QCD interactions. The study of their phenomenology (see e.g. De Luca et al. (2018) [803]) shows that they are viable DM candidates for multi-TeV masses: the direct detection cross section $\sigma \sim \Lambda_{\mathrm{QCD}}^4/M_Q^6$ satisfies the present bounds for $M_Q \gtrsim 10\ \mathrm{TeV}$; the cosmological DM abundance is reproduced thermally for $M_Q \approx 12.5\ \mathrm{TeV}$ for a color octet; the indirect detection rates, dominated by recombination, e.g., $QQ + \bar{Q}\bar{Q} \to Q\bar{Q} + Q\bar{Q}$, are below the present bounds, while the collider bounds are satisfied for $M_Q \gtrsim (1-2)\ \mathrm{TeV}$.

   This kind of DM is allowed if, during the cosmological evolution, essentially only the bound states made of $Q$ are formed and the production of problematic bound states that contain ordinary light quarks is sufficiently suppressed. In the limit where all the interactions are much faster than the Hubble rate, the cosmological evolution leads to the production of states made out of only $Q$, because these have a bigger binding energy $E_B \gg \Lambda_{\mathrm{QCD}}$. Estimates of the relevant QCD rates suggest that the problematic hadrons have an abundance that is a few orders of magnitude smaller than the $Q$-only hadrons. Such a small abundance is allowed by the bounds on the strongly-interacting relics, as discussed in section 5.1.2. Furthermore, such states lead to an unusual DM signal: heavy hadrons that hit the upper atmosphere and then slowly sink, remaining in normal matter as rare heavy atoms, and could potentially be discovered in this way. Searches with excited tantalum, in particular, could probe them, since they can be sensitive to very small DM abundances (see Lehnert et al. (2020) in [187]).

### 9.3.3 Weakly Interacting Massive Particles

Weakly Interacting Massive Particles (WIMPs) are a class of particles that play a prominent role as DM candidates. Perhaps because of their prominence, the definition of a WIMP is not entirely clear-cut.

In the narrowest sense, WIMP refers to a particle that interacts with the $W$ and $Z$ electroweak bosons. Therefore, a DM candidate classified as a WIMP needs to be charged under the electro-weak part of the Standard Model gauge group, $SU(2)_L \otimes U(1)_Y$, have zero electromagnetic charge, be a color singlet, and be stable. If the WIMP has a weak-scale mass, $M \sim M_{W,Z} \sim 100 \, \mathrm{GeV}$ (usually interpreted loosely, up to several orders of magnitude), then the annihilation cross section of two WIMPs is comparable to what is required for the simplest thermal freeze-out mechanism to generate the observed cosmological DM abundance, see the discussion surrounding eq. (4.9) in section 4.1.1. In scenarios with multiple WIMPs, the lightest WIMP can be the Dark Matter, while co-annihilations may play an important role, see section 4.1.6.

When interpreted more broadly, however, the term WIMP is used to denote a stable particle, which couples to other weak-scale particles such as the Higgs or the top quark in the SM, has interactions *'as weak as'* the $SU(2)_L \otimes U(1)_Y$ gauge interactions of the Standard Model, and therefore also has a *'weak-scale-like'* annihilation cross section. Clearly, a much wider set of models falls under the WIMP moniker in this case, thus making the WIMP assumption less predictive. Although there is no longer a direct connection to the SM weak interactions, the WIMP mass is still typically taken to be around 100 GeV. For adequately chosen parameters the correct relic abundance is still set by thermal freeze-out, in which case the phenomenology of broadly defined WIMPs is similar to the WIMPs defined more literally.

Historically, the simplicity of the freeze-out production mechanism, and the fact that DM could be connected to the stabilization of the electroweak scale against quantum corrections (the solutions to the hierarchy problem, as discussed in the next chapter), made WIMPs stand out as a conceptually attractive DM candidate since the early 1980's. This focus on WIMPs was aligned with the focus on low energy supersymmetry, which during that period was the motivation for most of the beyond the SM searches. In fact, for many years WIMP and neutralino, the lightest neutral supersymmetric particle (see section 10.1.2), were colloquially essentially synonymous. The motivation was strong enough that it jump-started a large experimental program aimed at discovering WIMPs in either direct detection, indirect detection, or production at colliders.

So far, this large experimental program only resulted in bounds. While the WIMP idea is by no means excluded, the absence of a discovery did result in a shift and broadening of the experimental and theoretical focus in the last decade, which we extensively covered in the present review.

A specific realization of WIMPs, intended in the stricter sense given above, is the Minimal Dark Matter hypothesis, which is discussed in the following subsection. Many of the other WIMP candidates, that belong to either the stricter or the broader categories, such as neutralinos or extra-dimensional DM, are discussed in chapter 10.

### 9.3.4 Minimal Dark Matter

A DM model of the Minimal Dark Matter (MDM) type is a minimal extension of the SM where a single DM multiplet is added to the SM [804]. Many non-minimal models reduce to MDM in the limit where only one added field is relevant for DM. For instance, a supersymmetric wino (section 10.1.2) corresponds to an added triplet in the MDM context when it is 'pure', i.e. unmixed with other supersymmetric fermions. We will shortly mention several of such cases, and for that let us first consider the MDM set-up systematically.

The non-trivial multiplets charged under the SM gauge group $\mathcal{G}_{SM} = U(1)_Y \otimes SU(2)_L \otimes SU(3)_c$ that can be added to the SM and that contain elementary particles that can be acceptable DM candidates

| Quantum numbers | | | | DM could | $M_{\rm DM}$ in TeV | | $M_{\rm DM^\pm} - M_{\rm DM}$ | $\sigma_{\rm SI}$ in |
|---|---|---|---|---|---|---|---|---|
| $U(1)_Y$ | $SU(2)_L$ | $SU(3)_c$ | Spin | decay into | tree | non-pert | in MeV | $10^{-46}\,{\rm cm}^2$ |
| 1/2 | 2 | 1 | 0 | $EL$ | | 0.54 | 350 | $(0.4 \pm 0.6)\,10^{-3}$ |
| 1/2 | 2 | 1 | 1/2 | $EH$ | | 1.1 | 341 | $(0.3 \pm 0.6)\,10^{-3}$ |
| 0 | 3 | 1 | 0 | $HH^*$ | 2.0 | 2.5 | 166 | $0.23 \pm 0.04$ |
| 0 | 3 | 1 | 1/2 | $LH$ | 2.4 | 2.6 | 166 | $0.23 \pm 0.04$ |
| 1 | 3 | 1 | 0 | $HH, LL$ | 1.6 | ? | 540 | $0.001 \pm 0.001$ |
| 1 | 3 | 1 | 1/2 | $LH$ | 1.9 | ? | 526 | $0.001 \pm 0.001$ |
| 1/2 | 4 | 1 | 0 | $HHH^*$ | 2.4 | ? | 353 | $0.27 \pm 0.08$ |
| 1/2 | 4 | 1 | 1/2 | $(LHH^*)$ | 2.4 | ? | 347 | $0.27 \pm 0.08$ |
| 3/2 | 4 | 1 | 0 | $HHH$ | 2.9 | ? | 729 | $0.15 \pm 0.07$ |
| 3/2 | 4 | 1 | 1/2 | $(LHH)$ | 2.6 | ? | 712 | $0.15 \pm 0.07$ |
| 0 | 5 | 1 | 0 | $(HHH^*H^*)$ | 5.0 | 14 | 166 | $2.0 \pm 0.5$ |
| 0 | 5 | 1 | 1/2 | none | 4.4 | 14 | 166 | $2.0 \pm 0.5$ |

Table 9.2: **Minimal Dark Matter**. *The first columns define the quantum numbers of the possible DM weak multiplets. The following ones show: the possible decay channels into SM particles that need to be forbidden (those in parenthesis correspond to dimension-5 operators); the DM mass predicted from thermal abundance (either at tree level, left column, or including non-perturbative Sommerfeld and bound-state corrections, right column, the latter not computed in all cases); the predicted splitting between the charged and the neutral components of the DM weak multiplet; the prediction for the SI DD cross section $\sigma_{\rm SI}$.*

are listed in table 9.2. In order to be acceptable, DM must have a vanishing color and electric charge[11]

$$Q = T_3 + Y = 0, \tag{9.16}$$

where $Y$ is the hypercharge of the electroweak multiplet and $T_3$ the corresponding eigenvalue of the diagonal generator of $SU(2)_L$. The most prominent examples are discussed in detail:

$2_S$ The first possibility is the **inert Higgs doublet** model, also called the **Inert Doublet Model (IDM)**, where the SM field content is supplemented by the complex $H' = (h'^+, h'^0)$ field: a scalar weak doublet with $Y = 1/2$, which contains a complex neutral component $h'^0$ in its lower component, with $T_3 = -1/2$, and a charged component in its upper component, with $T_3 = -1/2$. An ad-hoc symmetry such as $H' \to -H'$ is needed to make it stable. The inert Higgs doublet $H'$ has the same gauge quantum numbers as the SM Higgs, $H$. In supersymmetry, this field would be called the *left-handed slepton* $\tilde{L}$. The phenomenology of inert Higgs doublet model is discussed in section 9.3.5.

$2_F$ The second possibility is a **higgsino-like** fermionic weak doublet with $Y = 1/2$. In supersymmetric models this field arises as a fermionic partner of the Higgs scalar. An ad-hoc symmetry is again needed in order to forbid the otherwise allowed renormalizable couplings to the SM fields, which, if present, would make the higgsino unstable.

$3_F$ Yet another possibility is a **fermionic weak triplet**, which can have $Y = 0, 1$. The $Y = 0$ case

---

[11]More precisely, the charge needs to be negligibly small from observation, see section 3.3.2: in the case of discrete choices, this leaves $Q = 0$ as the only option. Special colored multiplets can give allowed DM as QCD bound states, as discussed in section 9.3.2. The case of singlets of the SM gauge group has been considered in section 9.2.1 (scalar singlet) and 9.2.2 (fermion singlet), so they are not reported here.

corresponds to the mentioned **wino-like** DM [805], the fermionic partner of the $SU(2)_L$ vectors in supersymmetric models (section 10.1.2). In these models the wino-like DM limit arises when all supersymmetric interactions are irrelevant, and therefore all the interactions are dictated by the gauge quantum numbers. Also in this case, additional ad hoc symmetries, such as a $\mathbb{Z}_2$ or $U(1)_{B-L}$, are needed in order to forbid extra renormalizable couplings that would otherwise make the wino-like state unstable.

$3_S$ **Scalar triplets** with $Y = 0$ or 1 provide additional DM candidates, that, however, also need to be stabilised by invoking ad-hoc symmetries [804]. (The phenomenology of the $Y = 0$ candidate was also discussed in [806].)

4 Various **fermionic or scalar 4-plets** provide DM candidates, which yet again need ad-hoc stabilising symmetries. Even though the fermionic 4-plets can only decay via dimension-5 operators, this by itself does not suppress enough the decay rates, and additional symmetries need to be invoked even in this case.

$5_F$ The smallest multiplet that is automatically stable is a **fermionic quintuplet** (with one of the three possible $Y$ values, $Y = 0, 1, 2$). In this case all the interactions that can make the multiplet decay are forbidden by gauge and Lorentz symmetries, up to dimension-6 operators. The dimension 6 operators result in a slow enough decay rate, $1/\tau \sim M^5/\Lambda^4$, if the suppression scale is high, $\Lambda \sim M_{\rm Pl}$.

The list of phenomenologically interesting minimal DM candidates is finite, since the electroweak multiplets cannot be too larger in order not to result in a too fast renormalisation group running of the $SU(2)_L$ gauge coupling. Requiring perturbativity up to $M_{\rm Pl}$ means that the fermions can be at most in a quintuplet, while (real) scalars can be at most in a 6-plet. Table 9.2 therefore provides a relatively complete list of minimal DM candidates. Note that the 6-plets, not listed in table 9.2, are not automatically stable, and their phenomenology is similar to the 4-plets.

Since the fermionic 5-plet is automatically stable, without requiring extra symmetries, and with no extra parameters apart from the DM mass, it is often referred to as *the Minimal DM* candidate. However, as mentioned at the beginning of the section, the notion of MDM is wider, and in general refers to a group of DM candidates with a similar phenomenology, rather than to a single DM candidate. Importantly, in its most constrained version the MDM mass is fixed from the observed thermal abundance (see below), which means that the MDM model is fully predictive, without free parameters.

We now move to the phenomenology of MDM candidates. The full renormalizable Lagrangian of the model is given by (here $\mathcal{X}$ denotes the added multiplet):

$$\mathscr{L} = \mathscr{L}_{\rm SM} + c \begin{cases} \bar{\mathcal{X}}(i\slashed{D} + M)\mathcal{X} & : \mathcal{X} \text{ is a spin } 1/2 \text{ fermionic multiplet,} \\ |D_\mu \mathcal{X}|^2 - M^2|\mathcal{X}|^2 & : \mathcal{X} \text{ is a spin } 0 \text{ bosonic multiplet,} \end{cases} \tag{9.17}$$

where $c = 1/2$ for a real scalar or a Majorana fermion and $c = 1$ for a complex scalar or a Dirac fermion. DM phenomenology is controlled by the gauge interactions of the $\mathcal{X}$ multiplet. These arise from the covariant derivative $D$, and are fully determined by the gauge quantum numbers. If $\mathcal{X}$ is a fermionic multiplet, then the multiplet mass $M$ is the only free parameter, which, furthermore, is determined by the relic abundance (see below), and thus all DM physics is fully predicted. If $\mathcal{X}$ is a scalar multiplet, extra quartic couplings are allowed,[12] and therefore the scalar case is 'less minimal' (unless the quartic

---

[12]Explicitly, they consist of $\mathscr{L}_{\rm non\ minimal} \supset -c\,\lambda_H(\mathcal{X}^*T^a_\mathcal{X}\mathcal{X})(H^*T^a_H H) - c\,\lambda'_H|\mathcal{X}|^2|H|^2 - \lambda_\mathcal{X}(\mathcal{X}^*T^a_\mathcal{X}\mathcal{X})^2/2 - \lambda'_\mathcal{X}|\mathcal{X}|^4/2$. Specific representations allow for extra quartic terms, such as the one relevant for the inert doublet model discussed in section 9.3.5.

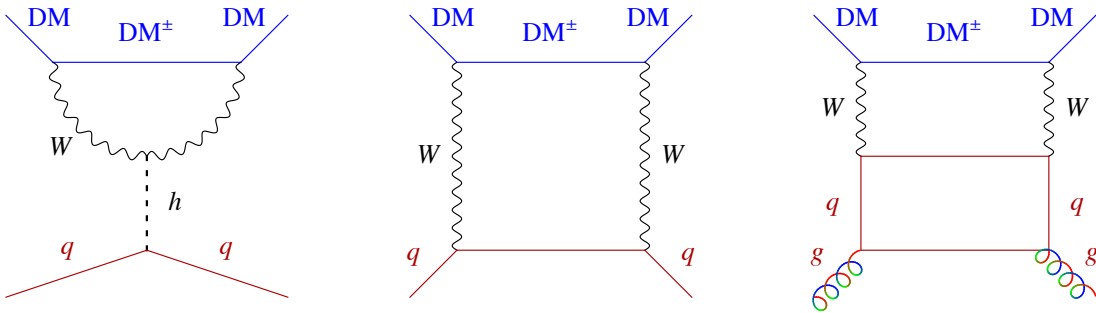

Figure 9.4: *The main 1-loop and 2-loops Feynman diagrams relevant for direct detection of fermionic Minimal Dark Matter with $Y = 0$.*

couplings are assumed to be negligible). This can be contrasted, for instance, with supersymmetry, which predicts many multiplets and the corresponding interactions among them. However, as mentioned above, in the limit where the lightest supersymmetric particle is much lighter than the other states, its phenomenology reduces to the one of MDM. Such a limit is sometimes referred to as a 'hot-spot', a predictive point in the otherwise large parameter space. In this sense, MDM candidates are hot-spots of more variegated theories.

## Mass splitting

At tree-level, all the components of the multiplet $\mathcal{X}$ have the same mass $M$. Since $SU(2)_L$ is broken, however, quantum corrections split the components with electric charges $Q$ and $Q'$, so that at one loop we have

$$M_Q - M_{Q'} = \frac{\alpha_2 M}{4\pi} \left\{ (Q^2 - Q'^2)s_W^2 f\left(\frac{M_Z}{M}\right) + (Q - Q')(Q + Q' - 2Y)\left[f\left(\frac{M_W}{M}\right) - f\left(\frac{M_Z}{M}\right)\right] \right\}.$$
(9.18)

The loop function equals $f(r) \overset{r \to 0}{\simeq} 2\pi r$ in the limit $M \gg M_{W,Z}$ [804]. In this limit the mass splitting can be understood as simply the Coulomb energy stored in the electric fields that surround each component of the multiplet. Such energy gives an additional contribution to the mass of a point particle. For a point-like particle of any spin, which couples with coupling $g$ to a light abelian vector of mass $M_V$, one therefore has

$$\delta M = \int d^3 r \left[\frac{1}{2}(\boldsymbol{\nabla}\varphi)^2 + \frac{M_V}{2}\varphi^2\right] = \frac{\alpha}{2}M_V + \infty, \quad \text{where} \quad \varphi(r) = \frac{g}{4\pi r}e^{-M_V r}.$$
(9.19)

Summing the contributions of the $\gamma, Z, W$ gauge bosons gives eq. (9.18). Numerically, the mass splitting between DM and the next lightest state is $M_{Q=1} - M_{Q=0} = 166\,\text{MeV}$, for $Y = 0$.

## Direct detection of Minimal DM

All MDM candidates with $Y \neq 0$ are excluded because they have a vector-like interaction with the $Z$ boson with an order one coupling, $-g_2 Y / \cos\theta_W$. At tree level, this produces a too large spin-independent

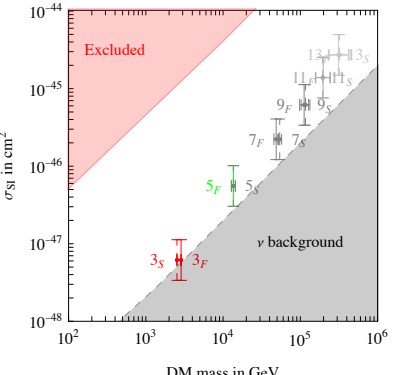
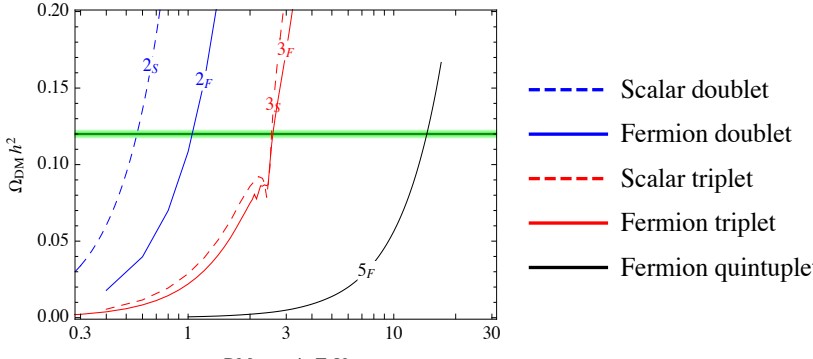

Figure 9.5: *Minimal Dark Matter in an n-plet representation of the* $\mathrm{SU}(2)_L$ *gauge group, denoted as* $n_{F,S}$, *where scalars (S) are dashed and fermions (F) continuous.* **Left**: *predictions for the mass and direct-detection cross section (from Bottaro et al. in [804]).* **Right**: *thermal DM abundance as a function of DM mass; the* $3_F$ *and* $5_F$ *curves include effects due to the bound states.*

elastic cross section on nuclei, see section 5.1.4,

$$\sigma_{\mathrm{SI}}^A = c\frac{G_{\mathrm{F}}^2 m_A^2}{2\pi}Y^2(N - (1 - 4s_{\mathrm{W}}^2)Z)^2, \tag{9.20}$$

where $c = 1$ for fermionic DM and $c = 4$ for scalar DM, while $N$ and $Z$ are the neutron and proton content of the nucleus, respectively. This cross section is a few orders of magnitude above the bounds from direct detection searches for $M \sim$ TeV, see section 5.

The MDM candidates with $Y \neq 0$ are still allowed in non-minimal set-ups where the complex component of $\mathcal{X}$ with $Q = 0$ is slightly mass split. The two resulting real components cannot have vector interactions, thus avoiding direct detection bounds. For an inert Higgs $H'$ this can be achieved trough a renormalizable $(H'H^*)^2$ quartic coupling (section 9.3.5). For a fermionic Higgsino this can be achieved by adding an extra real massive fermionic multiplet so that one can write down a Yukawa coupling with the SM Higgs; the extra fermion is either an electroweak singlet or a triplet (the latter possibility is automatically present in supersymmetric models).

Direct detection of MDM candidates with $Y = 0$ (and of the MDM candidates with $Y \neq 0$ in the non-minimal set-ups) proceeds via loop diagrams shown in fig. 9.4. The scattering cross sections are the same for the case of either boson or fermion MDM, and are suppressed by $1/M_W$ and $1/M_h$, rather than by $1/M$. The reason is that the scattering is dominated by the cloud of electro-weak vectors that surrounds the heavy DM particle. In some cases, the accidental cancellations between the Higgs and the $W$-mediated one loop diagrams requires the computation of two-loop diagrams with external gluons. The final results for the spin-independent cross sections are listed in table 9.2 and plotted in fig. 9.5 (left): they are below the present bounds and above the neutrino fog. A future 50 ton scale Xenon detector can thus exclude or discover all of the real MDM candidates ($Y = 0$) and severely constrain the parameter space of complex MDM candidates with nonzero hypercharge (see Bloch et al (2024) in [804]).

### Cosmological relic abundance of Minimal DM

DM annihilation cross section, relevant for thermal freeze-out, needs to be summed over all quasi-degenerate Minimal DM components, in order to take co-annihilations into account. The $s$-wave (co-

)annihilation cross section at tree level, in the normalisation of section 6, is given by

$$
\sigma v_{\rm rel} =
\begin{cases}
\dfrac{g_2^4(3 - 4n^2 + n^4) + 16Y^4 g_Y^4 + 8g_2^2 g_Y^2 Y^2(n^2 - 1)}{128\pi M^2\, g_{\mathcal{X}}} & \text{if } \mathcal{X} \text{ is a scalar,} \\[2ex]
\dfrac{g_2^4(2n^4 + 17n^2 - 19) + 4Y^2 g_Y^4(41 + 8Y^2) + 16g_2^2 g_Y^2 Y^2(n^2 - 1)}{256\pi\, M^2\, g_{\mathcal{X}}} & \text{if } \mathcal{X} \text{ is a fermion,}
\end{cases}
\tag{9.21}
$$

where an $n$-plet of $SU(2)_L$ has $g_{\mathcal{X}} = 2n$ degrees of freedom, if it is a complex scalar, while $g_{\mathcal{X}} = n$ for a real scalar, and $g_{\mathcal{X}} = 4n(2n)$ for a Dirac (Majorana) fermion. Annihilations into SM fermions and Higgs are $p$-wave suppressed, if $\mathcal{X}$ is a scalar.

These expressions for the annihilation cross sections receive in the non-relativistic regime non-perturbative corrections, if $M \gtrsim M_W/\alpha_2$: the Sommerfeld corrections and the bound-state formation (see section 4.1.5). Table 9.2 lists for each choice of the Minimal DM multiplet the mass $M$ for which the cosmological DM abundance is reproduced, both in the tree-level approximation as well as when non-perturbative corrections are included. The required values of DM mass $M$ are in the TeV to multi-TeV range, somewhat above the reach of the present colliders.[13]

## Indirect detection of Minimal DM

The fermionic MDM candidates with $Y = 0$ mainly annihilate into $W^+W^-$ at tree level, while annihilations into $\gamma\gamma$ and $\gamma Z$ arise at 1 loop, see fig. 9.6. Scalar MDM candidates can, in addition, also annihilate into pairs of Higgs bosons.

As for any other DM candidates, the annihilations of MDM particles result in various species of cosmic rays: $e^+$, $\bar{p}$, $\bar{d}$, $\gamma$. The phenomenology of thus generated charged cosmic rays follows the general discussion in section 8.2.8. The prompt gamma ray spectrum, however, is somewhat peculiar, since it features a prominent line at $E \simeq M$ in addition to a continuum shoulder at $E < M$. The latter is formed by $\gamma$-rays from the gauge bosons created in the final state parton showers. The $\gamma$ ray peak at $E \simeq M$, on the other hand, is generated at 1-loop. In most models it is suppressed relative to the continuum contribution, but not in Minimal DM. This non-trivial result is a consequence of the Sommerfeld enhancement (see section 4.1.5), which leads to comparable peak and continuum annihilation cross sections, for a range of DM masses.

The current status of the constraints for the most popular candidates (the *wino-like 3-plet* and *the Minimal DM 5-plet*) depends significantly on the assumed DM density profile in the Milky Way: if DM is cuspy in the center of the Galaxy, such candidates are ruled out across a large range of masses by the HESS limits, including the DM mass implied by the observed cosmological DM relic density. These models instead remain mostly viable if the DM density profile is non-cuspy.

In many DM models, DM particles that get captured in the Sun and then annihilate into neutrinos lead to important constraints, see section 6.9. This is not the case for the Minimal DM models. Given the small values of the scattering cross sections on nuclei, in MDM models the capture in the Sun is very inefficient: the prospects for detecting a flux of high energy neutrinos from the Sun's core that would be due to MDM annihilations are bleak.

---

[13] Minimal DM is the minimal realization of the 'WIMP miracle', i.e., that thermal relic DM with weak interactions has a roughly weak scale mass, see sections 9.3.3 and 4.1.1. In the past, the 'WIMP miracle' was often understood as pointing to $M \approx 100\,\text{GeV}$, as this resonated with the low energy supersymmetry hypothesis, which for such low WIMP mass would have also explained the small value of the Higgs mass, i.e., this would have been a natural solution to the hierarchy problem. MDM shows how in DM models with minimal extra content the 'WIMP miracle' is actually realized for larger, multi-TeV, values of the DM mass. This is, in particular, a consequence of the larger size of the multiplets (co-annihilations are important and thus the cross section larger, see the high powers with which $n$ enters in eq. (9.21)) and of Sommerfeld and bound-state enhancements.

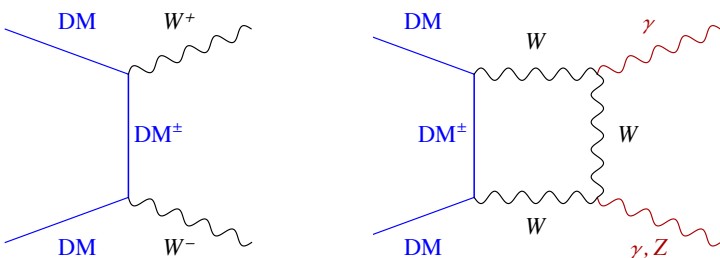

Figure 9.6: *Illustration of the main tree-level and 1-loop diagrams relevant for indirect detection of fermionic Minimal Dark Matter with $Y = 0$.*

## Collider searches for Minimal DM

If the MDM particles are lighter than what is required to match the observed cosmological DM density via the minimal thermal freeze-out (that is, if cosmology is more complicated), then the MDM can be searched for at the LHC, via the standard signal channels: the mono-jet, the mono-photon, the vector boson fusion processes, and others (see section 7). A peculiarity of this class of models is the presence of slightly heavier charged states $\mathcal{X}^\pm, \mathcal{X}^{\pm\pm}$. When the charged components of the MDM multiplet are produced in the LHC collisions, they decay into the neutral component and very soft charged pions, which are not reconstructed at the LHC. However, due to the smallness of the mass splitting, their lifetimes are relatively long; the corresponding decay lengths at rest are on the order of a few centimeters. A small but non-negligible fraction of the charged states, corresponding to the tail of the decay distribution, would therefore travel far enough to leave a track in the detector. These events would appear as high $p_T$ charged tracks that suddenly disappear inside the detector (the act of disappearing occurs when the charged state decays into DM and the unreconstructed $\pi^\pm$).

Current searches at the LHC are sensitive at most to Minimal DM masses of a few hundred GeV. MDM candidates provide, however, a well defined target for future colliders: the interactions of MDM are fully predicted by their quantum gauge numbers, while a thermal freeze-out indicates multi-TeV values for their masses. Disappearing charged tracks would provide maximal sensitivity, if these will be observable by tracker components of the future detectors. In such an optimistic case, it is estimated that a futuristic 100 TeV $pp$ collider could directly probe a (Higgsino-like) fermion doublet up to $\sim 1.5\,\mathrm{TeV}$, a (Wino-like) fermion triplet up to $\sim 6\,\mathrm{TeV}$, and a scalar triplet up to $\sim 3\,\mathrm{TeV}$, thereby covering the masses predicted by the thermal freeze-out [807]. However, a fermion 5-plet would be probed only up to $\sim 6\,\mathrm{TeV}$, not covering the mass predicted by the thermal freeze-out. This unfortunately provides a counter-example to the often repeated claim that a 100 TeV $pp$ collider can conclusively probe weakly interacting dark-matter particles of thermal origin [808]. A futuristic muon collider could almost reach its kinematical limit $M \lesssim \sqrt{s}/2$, and could thus cover all MDM thermal targets for $\sqrt{s} \sim 30\,\mathrm{TeV}$ [807].

## DM and neutrino masses

Minimal DM candidates can be part of extended models that generate neutrino masses at one loop. For example, a model is obtained adding to the SM the MDM fermion 5-plet $\mathcal{X}$ with hyper-charge $Y = 0$ and a scalar 4-plet $S$ with $Y = 1/2$. The resulting theory is invariant under $\mathcal{X} \to -\mathcal{X}$ and $S \to -S$ and contains a Yukawa coupling to leptons $L\mathcal{X}S$ and a $S^2H^2$ quartic. Together with the Majorana mass of $\mathcal{X}$, these interactions break lepton number. Then, at one loop level $\mathcal{X}$ and $S$ mediate the Majorana neutrino mass operator $(LH)^2$ [809].

### 9.3.5  Inert Higgs

The special case of an extra scalar doublet $H'$ with vanishing Yukawa couplings and a vanishing vacuum expectation value is known as the *'inert Higgs'* or the *'Inert Doublet Model (IDM)'* [810]. In order to enforce the stability of the neutral component in $H'$, and thus a viable DM candidate, an unbroken $H' \to -H'$ symmetry is assumed (under which all the SM fields, including the usual SM higgs doublet, are even). Particularly relevant is the quartic term $\lambda_5 \mathrm{Re}\,(H^* H')^2$ in the scalar potential, which splits the inert Higgs into its components, $H' = (h'^+, (h' + i\eta')/\sqrt{2})$, and qualitatively differentiates an Inert Higgs from a Minimal DM scalar doublet. The DM candidate is the lightest neutral state, which is either the scalar $h'$ or the pseudo-scalar $\eta'$, depending on the sign of the $\lambda_5$ coupling. This mass splitting avoids exclusion from direct detection searches, because the $Z$-mediated scattering is now inelastic, see section 5.5.1. The Higgs-mediated scattering amplitude due to $\lambda_5$ and other scalar couplings leads to small scattering rates, which are below the current constraints but can be searched for in the future. The DM cosmological abundance is reproduced via thermal freeze-out in two main distinct regions:

1. A 'high mass' region around $M \sim 500\,\mathrm{GeV}$ where annihilations are dominantly into vector bosons, like in the $\mathrm{SU}(2)_L$-invariant Minimal DM limit, see eq. (9.17). Thereby this region is the only one considered in table 9.2.

2. Some 'low-mass' islands around $M \sim 70\,\mathrm{GeV}$, where the dominant DM annihilation process is mediated by the SM Higgs in the $s$-channel.

Dedicated analyses find that even the second region is currently allowed by collider constraints.

## 9.4  Models with dark forces and dark sectors

We now move to the scenarios in which DM is not charged under the known SM forces but rather charged under one or more hypothetical *dark force(s)*. The simplest possibility is a dark U(1) gauge symmetry. The associated dark gauge boson can be much lighter than the DM itself; it can even be massless and result in a long range dark interaction. The presence of a dark force changes significantly the phenomenology of dark matter. For instance, the DM DM annihilations in the presence of a long range dark force are Sommerfeld enhanced (see section 4.1.5) and, in general, DM will be self-interacting (see section 3.3.3). In more general models, the DM particle could be accompanied by a whole "dark sector" of particles, possibly lighter than the DM itself. Such dark sectors can lead to a large set of possible signals that are accessible at energies within reach of laboratory experiments, see section 7.4.

That DM could be accompanied by a much larger dark sector should come as no surprise, as it would be reminiscent of the situation in the visible sector where the two stable matter particles, electron and proton, are accompanied by a much larger structure of elementary particles. Since the dark sector could be as complicated or even more complicated than the SM sector, it is next to impossible to cover all possible non-minimal models. One possibility is to focus on a particular complicated structure, hoping that this covers a large enough set of signatures. A popular choice is to postulate that the dark sector consists of one or more copies of the SM [771]. This will be discussed in subsection 9.5.3. Another option is to considering dark sectors that are obtained by spontaneous breaking of a 'unified' dark gauge symmetry, a possibility that we discuss in subsection 10.6.

In the bulk of this section we follow a more minimal organizing principle, and assume that interactions between the SM and the dark sector can be well approximated as [697]

$$\mathscr{L}_{\mathrm{portal}} = \sum_i c_i \mathcal{O}_{i\mathrm{SM}} \mathcal{O}_{i\mathrm{DS}} \,, \tag{9.22}$$

| Portal | Interactions |
|---|---|
| Dark Photon, $A'_\mu$ | $-\epsilon F'_{\mu\nu} B^{\mu\nu}$ |
| Dark Higgs, $S$ | $(\mu S + \lambda S^2) H^\dagger H$ |
| Heavy Neutral Lepton, $N$ | $y_N L H N$ |
| Axion-like pseudo scalar, $a$ | $a F \tilde{F}/f_a,\ a G \tilde{G}/f_a,\ (\bar{\psi}\gamma^\mu\gamma_5\psi)\partial_\mu a/f_a$ |

Table 9.3: *The lowest dimension portals to the dark sector, and sketches of the corresponding interactions (see the main text for precise $\mathcal{O}(1)$ factors).*

where $\mathcal{O}_{\mathrm{SM}}$ and $\mathcal{O}_{\mathrm{DS}}$ are local operators in the SM and dark sectors, respectively. The most relevant *portal interactions* are those that are minimally suppressed by some high-energy scale. Their structure depends on the unknown quantum numbers of dark sector particles, usually dubbed a *"dark Higgs"* or a *"light singlet"*, if the dark sector particle is a spin 0 scalar; an *"axion"* or *"axion-like pseudo-scalar*, if it is a spin 0 pseudo-scalar; a *"dark photon"* or a *"light Z′"*, if it is a spin 1 vector; or a *"heavy neutral lepton"*, if it is a spin 1/2 fermion. At lowest mass dimensions of interactions, i.e., at dimensions 4 and 5, only a limited number of portals $\mathscr{L}_{\mathrm{portal}}$ can be written down. These are listed in table 9.3. The dark photon, dark Higgs, and the neutral lepton portals are renormalizable, while the axion portal is suppressed by a high-energy scale represented by the axion decay constant, $f_a$, since it is assumed to be a pseudo-Nambu-Goldstone boson.

In most dark sector constructions, and therefore also in this section, the common assumption is that the dark sector particles discussed above are merely mediators between the dark and visible sectors. The dark matter particle is usually assumed to be a different, additional state in the dark sector. However, a more minimal option is also possible — the mediator itself, if sufficiently stable, can be the DM candidate, as long as it is produced in the correct amount to match the cosmological observations. Dark photon dark matter is briefly addressed at the end of subsection 9.4.1 below, as well as in section 9.8; axion-like particles as the DM are discussed in sections 3.4 and 10.4; heavy neutral lepton (a.k.a. sterile neutrino) as the DM is discussed in section 9.2.2; dark Higgs (scalar singlet) as a DM is discussed in section 9.2.1.

## 9.4.1   The vector portal

The dark photon or vector portal couples the dark sector with the SM through a kinetic mixing term [710],

$$\mathscr{L}_{\mathrm{dark\ photon}} = \mathscr{L}_{\mathrm{SM}} + \mathscr{L}_{\mathrm{DS}} + \mathscr{L}_{\mathrm{mix}}, \qquad \mathscr{L}_{\mathrm{mix}} = -\frac{\epsilon}{2\cos\theta_{\mathrm{W}}} F'_{\mu\nu} B^{\mu\nu}, \qquad (9.23)$$

where $B_{\mu\nu} = \partial_\mu B_\nu - \partial_\nu B_\mu$ and $F'_{\mu\nu} = \partial_\mu A'_\nu - \partial_\nu A'_\mu$ are the field strengths of the SM hypercharge, $\mathrm{U}(1)_Y$, and the new dark $\mathrm{U}(1)'$, respectively. The weak angle $\theta_{\mathrm{W}}$ is kept in the definition of $\epsilon$ so that, after the electroweak symmetry breaking, the kinetic mixing term becomes

$$\mathscr{L}_{\mathrm{mix}} = \frac{1}{2}\epsilon F'_{\mu\nu}\big(F^{\mu\nu} - \tan\theta_{\mathrm{W}} Z^{\mu\nu}\big), \qquad (9.24)$$

where $F_{\mu\nu}$ ($Z_{\mu\nu}$) are the SM photon ($Z$) field strengths. The kinetic mixing parameter, $\epsilon$, is generated at least radiatively. If there are heavy fields charged under both $\mathrm{U}(1)_Y$ and $\mathrm{U}(1)'$, the mixing is generated at 1-loop, $\epsilon \sim g_Y g_D/(4\pi)^2$. If not, the kinetic mixing is generated at higher loops unless it is explicitly forbidden by a symmetry (see also the discussion in section 3.3.2).

The dark photon Lagrangian may also contain a non-trivial dark sector with extra dark sector matter

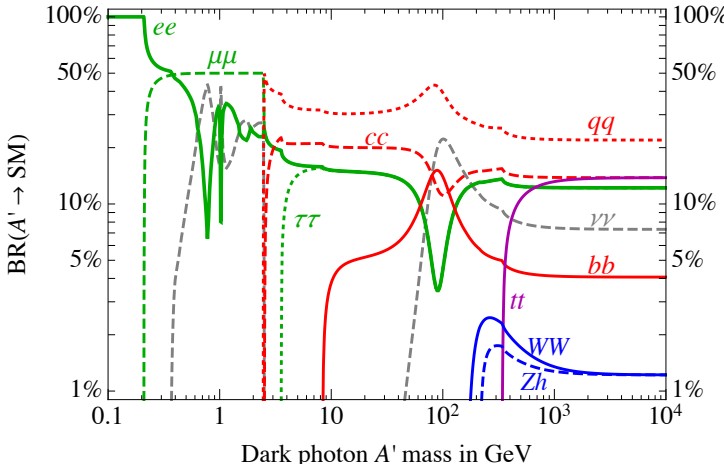

Figure 9.7: *Predicted **branching ratios for dark photon decays** with mass $M_{A'}$. The $A' \to e^+e^-$ decay mode (green solid curve) dominates below the muon threshold, while above it $\mathrm{BR}(A' \to \mu^+\mu^-)$ (green dashed) catches up. The decays into hadronic channels open around the $\pi^+\pi^-$ threshold. Between $350\,\mathrm{MeV} \lesssim M_{A'} \lesssim 2.5\,\mathrm{GeV}$ the hadronic resonances are approximated to decay half into $\mu^+\mu^-$ and half into $\nu\bar{\nu}$ (this approximation misses a small fraction of other SM particles, like photons from $\pi^0$ decays in the final state). At $M_{A'} \gtrsim 2.5\,\mathrm{GeV}$, the decay into quark pairs is perturbatively described and these channels pick up. At larger masses, different decay channels open at the relevant thresholds. (Figure adapted from Sala et al. (2018); see therein for details. For alternative presentations where the hadronic resonances are not decayed, see [811]).*

fields, $\chi$. Assuming for concreteness that $\chi$ is a Dirac fermion we have

$$\mathscr{L}_{\mathrm{DS}} = -\frac{1}{4}\left(F'_{\mu\nu}\right)^2 + \frac{1}{2}M_{A'}^2\left(A'_\mu\right)^2 + \bar{\chi}(i\partial_\mu - g_D A'_\mu)\gamma^\mu\chi - m_\chi\bar{\chi}\chi + \cdots. \qquad (9.25)$$

The dark fermion $\chi$ could be the DM, but does not have to be. The dark photon mass $M_{A'}$ is most often taken as a free parameter since only tree level effects are usually considered. If a UV complete model is needed, $M_{A'}$ can be assumed to come from spontaneous symmetry breaking, in which case $\mathscr{L}_{\mathrm{DS}}$ also contains a dark Higgs charged under $\mathrm{U}(1)'$. The dark Higgs obtains a vacuum expectation value, which then breaks the $\mathrm{U}(1)'$.

The dark photon phenomenology is most easily derived in a basis where the quadratic part of the action is diagonal. If the dark photon is massive, the mixing with the photon can be removed by a redefinition of the photon field

$$A_\mu \to A_\mu + \epsilon A'_\mu, \qquad (9.26)$$

while the $A'_\mu$ field remains unchanged (and thus the $A'$ mass term is trivially still diagonal). As a consequence of the redefinition of eq. (9.26), the massive hidden photon couples to all electrically charged particles with the strength $\epsilon e$, and can be produced in the collisions of SM particles. At low energies the $Z$ boson can be ignored.[14]

The phenomenology of the vector portal depends on the details of the dark sector. In the **minimal dark photon model** the dark photon is assumed to be the lightest dark sector state, while the other

---

[14]Including the $Z$ boson in the diagonalization, the new interactions are up to $\mathcal{O}(\epsilon^2)$ given by the interaction Lagrangian $-e\epsilon J^\mu A'_\mu + g_D\epsilon\tan\theta_{\mathrm{W}}J'^\mu Z_\mu + g_D J'^\mu A'_\mu$, where $J^\mu$ ($J'^\mu$) is the SM electromagnetic current (the dark current).

dark sector particles are heavier and can be ignored. The phenomenology of the minimal dark photon model is controlled by only two parameters, $M_{A'}$ and $\epsilon$. If the dark photon is heavy enough, it decays back to the SM particles through the $-e\epsilon J^\mu A'_\mu$ interaction, leaving a visible imprint in the detector. All the relative branching ratios, $A' \to e^+e^-, \mu^+\mu^-, \pi^+\pi^-$, etc., are fixed by the electro-magnetic current $J^\mu = -\left(\bar{e}\gamma^\mu e\right) + \frac{2}{3}\left(\bar{u}\gamma^\mu u\right) - \frac{1}{3}\left(\bar{d}\gamma^\mu d\right) + \cdots$, making the model predictive, see fig. 9.7.[15] A dark photon lighter than $2m_e$ does not decay into visible SM states and instead appears as missing energy in the detector (such a light $A'_\mu$ does decay to SM neutrinos through one loop induced electroweak transition, with the neutrinos escaping the detector). A massless dark photon leads to the effects and constraints discussed around eq. (9.35).

A less minimal option is **light dark matter coupled to dark photon**. The DM $\chi$ in eq. (9.25) is then taken to be lighter than the dark photon so that the invisible decays $A' \to \chi\bar{\chi}$ are possible. In this non-minimal option the parameter space becomes larger, but still tractable, with four unknown parameters $\epsilon, g_D, M_{A'}, m_\chi$. For $g_D \gg \epsilon e$ the invisible decay channel dominates and the bounds on vector portal become much weaker, compared to the dark photon model.

Finally, the **millicharged particles** limit is obtained when $M_{A'} \to 0$ and both $U(1)_{em}$ and $U(1)'$ are unbroken. The absence of vector masses allows to freely rotate the $A_\mu, A'_\mu$ fields. This can be used to choose a basis more convenient than eq. (9.26), imposing that SM charged particles couple to one vector only, the photon. The needed transformation is

$$A'_\mu \to A'_\mu - \epsilon A_\mu, \qquad A_\mu \to A_\mu, \tag{9.27}$$

In this basis the DM $\chi$ carries a millicharge $|Q_\chi| \simeq |\epsilon g_D e|$ under the visible photon.[16] The general phenomenology of millicharged DM is discussed in section 3.3.2.

More generally, if the SM fermions are charged directly under the $U(1)'$, the dark photon is usually renamed as $Z'$. An example of such a $Z'$ is, for instance, the gauge boson of $U(1)_{B-L}$ under which all the SM quarks carry a charge $+1/3$ and the SM leptons carry a charge $-1$ such that $U(1)_{B-L}$ is anomaly free already just with the SM fermion content. Other options are possible, such as gauging the $L_\mu - L_\tau$ number, etc. The experimental bounds on dark photons or $Z'$s with general couplings can be obtained through the `DarkCast` code [812].

The experimental constraints for the minimal dark photon model and for the dark matter coupled to a dark photon are shown in fig. 7.5.

## Dark Photon Dark Matter

As already mentioned above, the dark photon itself can be the dark matter [813]. A number of different mechanisms for the production of the dark photon cosmological abundance have been considered in the literature, including the misalignment mechanism (see section 4.3.4) as well as the inflationary production (section 4.3). The plausible mass range for dark photon DM is wide, since it depends on the production mechanism. Typically, it extends from 1 MeV on the high end, down to the minimum DM mass value of about $10^{-21}$ eV (see eq. (3.1)), although in specific models dark photon masses as large as $10^{14}$ GeV were also considered.

The search strategies for dark photon dark matter are in broad strokes rather similar to the ones for the axion searches (though some of the techniques are specific to axions, see section 10.4.5). For

---

[15]We acknowledge the help of F. Sala for the figure.

[16]The physics of course does not depend on which basis is used as long as both vector bosons are included in the calculations. For nonzero $M_{A'}$ the field redefinition eq. (9.27) does not keep the mass term diagonal, in contrast to eq. (9.26), making the latter a more convenient choice. The $M_{A'} \to 0$ limit is relevant when $q \gg M_{A'}$, where $q$ is the typical energy or momentum exchange in the process. For this to be valid in all observations, both at colliders ($q$ from $\approx 0.1\,\text{GeV}$ to $\approx$ TeV) and in astrophysics/cosmology, $M_{A'}$ needs to be very small, below $10^{-20}$ GeV.

a detailed review of dark photon DM experimental searches and resulting constraints see Caputo et al. (2021) in [813].

## 9.4.2   The scalar portal

A scalar singlet $S$ can couple to the SM Higgs bilinear, $H^\dagger H$, forming the so called *scalar* or *Higgs portal* to the dark sector [814],

$$\mathscr{L}_{\text{scalar}} = \mathscr{L}_{\text{SM}} + \mathscr{L}_{\text{DS}} - \left(\mu S + \lambda_{SH} S^2\right) H^\dagger H, \tag{9.28}$$

where $\mu$ ($\lambda_{SH}$) is a dimensionful (dimensionless) parameter. The set-up is very similar to that of scalar singlet DM (section 9.2) except that here $S$ is the mediator rather than the DM candidate, and thus there is no requirement of a $Z_2$ symmetry that would stabilize $S$. The dark sector Lagrangian is

$$\mathscr{L}_{\text{DS}} = \bar{\chi}(i\slashed{\partial} - m_\chi)\chi + \frac{1}{2}\left(\partial_\mu S\right)^2 - \frac{1}{2}M_S^2 S^2 + y_\chi \bar{\chi}\chi S + \cdots, \tag{9.29}$$

where for concreteness we again assume that $\chi$ is a Dirac fermion that may be the DM. After electroweak symmetry breaking, $H = (0, v + h/\sqrt{2})$ in the unitary gauge, the interaction term $\mu S H^\dagger H$ in (9.28) leads to the mixing between $S$ and the SM Higgs, $h$. The interactions of $S$ with the SM fields are induced through the mixing with the Higgs, and are thus the same as for the SM Higgs, except that they are suppressed by the mixing angle $\theta_S \simeq \mu v/(M_h^2 - M_S^2)$. Phenomenologically important are the loop induced $b \to sS$ and $s \to dS$ decays, inducing the meson decays, $B \to K^{(*)}S$ and $K \to \pi S$, respectively. The quartic coupling $\lambda_{SH}$ gives rise to $h \to SS$ decays, if $S$ is light enough, which can be a production channel of $S$ at the LHC. The singlet $S$ decays predominantly invisibly, if $2m_\chi < M_S$ and $y_\chi \gg \theta_S y_f$ (with $y_f = m_f/v$ the Yukawa coupling of the heaviest SM fermion the $S$ can decay to).

## 9.4.3   The neutrino portal

The dark sector is assumed to contain *heavy neutral leptons* (HNLs), i.e., gauge singlet fermions here denoted as Weyl fermions $N_i$ [815]. The HNLs are allowed by the SM gauge symmetry to have Yukawa interactions with the SM leptons, giving rise to the following portal[17]

$$\mathscr{L}_{\text{HNL}} = \mathscr{L}_{\text{SM}} + \bar{N}_i i\slashed{\partial} N_i - \left(\frac{M_{ij}}{2} N_i N_j + y_{ai} H N_i L_a + \text{h.c.}\right), \tag{9.30}$$

where the summation over the SM lepton labels, $a = 1, 2, 3$ and the HNL labels, $i = 1, \ldots, n_N$, is understood. The dark sector Lagrangian contains mass terms for HNLs, with both Majorana and Dirac mass terms allowed. Since the motivation for HNLs mainly comes from neutrino mass models, the origin and structure of these mass terms may be of interest in itself.

   After electroweak symmetry breaking the Higgs obtains a vacuum expectation value, and the Yukawa interaction in eq. (9.30) mixes the SM neutrinos, $\nu_a$, with the HNLs, $N_i$, resulting in the mass eigenstates $\tilde{\nu}_a, \tilde{N}_i$. The net effect is that the interaction of HNLs with the SM are obtained by simply substituting in $\mathscr{L}_{\text{SM}}$ the SM neutrinos with $\nu_a = \tilde{\nu}_a + U_{ai}\tilde{N}_i$, where $U$ is the mixing matrix. In the minimal HNL models both the production and the decays of HNLs are assumed to be fully determined by the mixing matrix elements $U_{ai}$ (in general, decays into the HNL sector may also be important). Very often an assumption of either electron, muon or tau dominance is made, i.e., that there is a single HNL, $N_1$, with only $U_{e1}$, $U_{\mu 1}$ or $U_{\tau 1}$ nonzero, respectively [697]. The HNL mass is taken as a free parameter.

   Unlike the sterile neutrino DM case, discussed in section 9.2.2, here there are no requirements for any of the HNLs to be the DM.

---

[17]Here and elsewhere we use the compact notation $\bar{N}_i i\slashed{\partial} N_i = N_i^\dagger \bar{\sigma}^\mu i\partial_\mu N_i$ for Majorana fermion kinetic terms.

### 9.4.4 The pseudo-scalar portal

The QCD axion is a light pseudo-scalar particle that is a pseudo Nambu-Goldstone boson (pNGB) arising from a spontaneous breaking of a global symmetry, solves the strong CP problem and is a viable dark matter candidate. The QCD axion is very light, with the axion mass due entirely to the QCD anomaly, see section 10.4 for further details.

The *axion like particle* (ALP) is instead a light pNGB that can be a viable DM candidate as a generalization of the QCD axion (see section 3.4) or can simply act as a mediator between the dark sector and the SM,

$$\mathscr{L}_{\text{ALP}} = \mathscr{L}_{\text{SM}} + \mathscr{L}_{\text{DS}} + \frac{a}{f_a}\frac{\alpha_{\text{s}}}{8\pi}G^a_{\mu\nu}\tilde{G}^{a\mu\nu} + c_\gamma\frac{a}{f_a}\frac{\alpha}{8\pi}F_{\mu\nu}\tilde{F}^{\mu\nu} + \sum_\psi c_\psi\frac{\partial_\mu a}{f_a}\big(\bar{\psi}\gamma^\mu\gamma_5\psi\big). \qquad (9.31)$$

Here the sum runs over the SM fermions, both quarks and leptons, and we kept only the flavor diagonal couplings. The ALP couples to the SM fermions via a derivative coupling, since it is a PNGB, while the non-derivative couplings of ALP to gluons and photons arise when the spontaneously broken global symmetry is anomalous with respect to QCD and $\text{U}(1)_{\text{em}}$. The ALP mass, $m_a$, is taken to be a free parameter, and from the low energy perspective represents an explicit breaking of the shift symmetry $a \to a + \alpha$, where $\alpha$ is a phase. (Note that the coupling to gluons also contributes to ALP mass, $m_a$. For the QCD axion this is the only contribution, see eq. (10.25) below).

The ALP parameter space is quite large. In addition to the ALP mass $m_a$, the coupling to gluons (given by $1/f_a$), and to photons ($c_\gamma/f_a$), there are also nine flavour-diagonal couplings to quarks and charged leptons ($c_\psi/f_a$), as well as possible flavour-violating couplings. Commonly used simplified benchmarks are: photon dominance (only $c_\gamma$ nonzero), gluon dominance (only the couplings to gluons are nonzero), or fermion dominance (taking for simplicity $c_\psi$ to be universal). This does give some estimate of how stringent the exclusions in the ALP parameter space are, but of course does not cover fully all the interesting possibilities. For instance, the flavor off-diagonal couplings to the SM fermions could well lead to the first observation of the ALP in the rare decays of $B, D$ or $K$ mesons [816]. Note also, that the axion portal interactions are non-renormalizable. Unlike the vector portal, the scalar portal and the neutrino portal, the axion portal thus requires a full high-energy theory, see section 10.4.3. The ALP could be a DM: the case where this is a QCD axion is discussed further in section 10.4.

## 9.5 Composite Dark Matter

Quite generally, particles can be either elementary or composite, and this holds true also for DM. Composite particles are made out of elementary particles (scalars, fermions or vectors)[18] and can be strongly bound, like quarks and gluons inside a proton, or weakly bound, like proton and electron forming a hydrogen atom. Typically, the compositeness scale $\Lambda$ corresponds to the inverse radius of the composite particle. Hence, at momentum exchanges smaller than $\Lambda$, and at energies smaller than the binding energy, composite DM appears as a point particle. This means that, for such low energies, our discussion so far, in which we treated DM as a point particle, still applies. However, the renormalizable DM interactions get supplemented by additional non-renormalizable operators suppressed by the compositeness scale $\Lambda$ (sometimes these can be resummed into DM form factors, as was done for nuclear physics in the case of DM scattering on nuclei, see sections 5.1.4 and 5.1.5, but now on the DM side). This means that, for precise phenomenology or for momentum exchanges above the compositeness scale, the composite nature of DM needs to be taken into account. If a DM signal will be observed in the future, one can try to

---

[18]We do not discuss in detail the AdS/CFT duality [817] as a tool for approximating near-conformal strong interactions, via models in warped extra dimensions, irrespectively of what the constituents are. For general discussion of extra dimensional DM models see section 10.1.3.

measure these non-renormalizable interactions, and then use them to reverse-engineer the constituents and identify the right microscopic model.

The interplay between the compositeness scale and the constituents' mass also controls the DM mass. In strongly coupled theories when the masses of the constituents are smaller than the compositeness scale, then $M \approx \Lambda$. An example is the mass of a proton in the SM, which is comparable to the non-perturbative scale of QCD, $\Lambda_{\rm QCD} \approx 1\,{\rm GeV}$. If constituents are heavier than the compositeness scale, the DM mass is instead $M \gg \Lambda$. A SM example is the hydrogen atom, whose mass is to a good approximation given simply by the sum of the constituent masses, $m_{\rm H} \approx m_p + m_e$. This is much larger than the compositeness scale, corresponding in this case to the inverse Bohr radius, $1/a_0 = \alpha m_e$.

At the moment, there are many possibilities for the models of composite DM. Dark sectors can be quite complex in general, in the same way that the visible sector has a complex structure in the SM; with confining dynamics, long range forces, several stable states, etc. This means that, even in simple theories, stable dark relics could also follow a rather complex production history, quite different from the simplest thermal relic examples discussed in chapter 4. In fact, one can assume that the origin of dark matter may be linked to the origin of visible matter, i.e., that the generation of baryon asymmetry (the dominance of visible matter over antimatter) may be linked to the generation of dark baryon asymmetry. Composite DM models are therefore often also models of asymmetric DM.[19] The latter is specifically discussed in sections 4.4 and 9.7. In this section we focus on compositeness models in general. Such models can be divided into strongly coupled ones (section 9.5.1) and weakly coupled ones (section 9.5.3). We will discuss their main features, such as the possible patterns of symmetry breaking and their general phenomenology, highlighting the differences with respect to elementary DM. A selection of concrete models can be found in [819].

## 9.5.1 QCD-like strongly coupled theories

Arguably, the best understood strongly coupled theory is QCD, where we have at our disposal many experimental and theoretical insights. It is thus not surprising that the most studied models of composite DM are QCD-like. These models assume as matter content $N_F$ flavours of dark fermions $\Psi_i$, $i = 1, \ldots, N_F$, that are in a vector-like representation of a 'Dark Color' gauge group $\mathcal{G}_{\rm DC}$, either $\mathrm{SU}(N)_{\rm DC}$, $\mathrm{SO}(N)_{\rm DC}$, or $\mathrm{Sp}(N)_{\rm DC}$, see table 9.4. Dark gauge interactions become non-perturbative at a scale $\Lambda$, resulting in a confinement of dark fermions and in the chiral symmetry breaking patterns listed in the second column of table 9.4.

Dark fermions $\Psi_i$ can in general be charged also under the SM gauge group, $\mathcal{G}_{\rm SM}$. The $N_F$ dark fermions are then organised into subsets, each of which belongs to a distinct irreducible representation of $\mathcal{G}_{\rm SM}$ (some of the dark sector states will thus have a nonzero electric charge). Like in QCD, one can get a complex spectrum of composite states, starting from a simple microscopic Lagrangian with only a few free parameters. An appealing possibility is that, as in the SM, the strongly coupled dark sector also exhibits an accidental global symmetry,[20] which then implies one or more stable composite particles, either dark baryons (composed of just $\Psi_i$) or dark pions (composed of $\Psi_i$ and $\bar{\Psi}_j$). These can be viable

---

[19]  Such composite DM can also form **asymmetric DM nuggets**, i.e., large composite states of $10^4$ or more asymmetric DM particles. Asymmetric DM models can lead to the formation of large bound states because they generically feature long-range attractive self-interactions due to a light mediator (these are needed so that the annihilation rate is large enough that it efficiently depletes the symmetric component) [818]. This same mediator provides the attractive force that leads to nuggets. See also the discussion of quark nuggets in the SM, section 9.1.2.

[20]A symmetry is *accidental* if it is not explicitly demanded in the construction of the renormalizable Lagrangian, but rather follows (at least in some limit) from the assumed field content and their charges under gauge interactions. For example, baryon number and lepton number arise as accidental global symmetries in the renormalizable limit of the SM.

DM candidates, if they are electrically neutral, which can be the case even if they are composed of charged constituents, in the same way as the neutron in the SM is composed of charged quarks.

## QCD-like dark baryons

An example of a theory where the DM consists of a dark baryon is a dark QCD with $N_F$ dark fermions in the fundamental representation of the Dark Color gauge group $\mathcal{G} = \mathrm{SU}(N)_{\mathrm{DC}}$. The renormalizable Lagrangian of the theory is

$$
\begin{aligned}
\mathscr{L} \;=\; & \mathscr{L}_{\mathrm{SM}} + \bar{\Psi}_i\big(i\slashed{D} - m_i\big)\Psi_i - \tfrac{1}{4}G^a_{\mathrm{DC},\mu\nu}G^{\mu\nu a}_{\mathrm{DC}} + \\
& + \frac{g^2_{\mathrm{DC}}}{32\pi^2}\theta_{\mathrm{DC}}G^a_{\mathrm{DC},\mu\nu}\tilde{G}^{\mu\nu a}_{\mathrm{DC}} + \big[H\bar{\Psi}_i\big(y^L_{ij}P_L + y^R_{ij}P_R\big)\Psi_j + \mathrm{h.c.}\big]
\end{aligned}
\tag{9.32}
$$

where $G^{\mu\nu a}_{\mathrm{DC}}$ is the DC gauge field strength, and $g_{\mathrm{DC}}$ the corresponding gauge coupling. The Yukawa couplings $y^{L,R}_{ij}$ of DC fermions with the SM Higgs $H$ are nonzero only when they are allowed by the SM and dark gauge symmetries. This Lagrangian is accidentally invariant under a $\mathrm{U}(1)_{\mathrm{DC}}$ global symmetry, which rotates all the DC fermions $\Psi_i$ by the same phase, $\Psi_i \to e^{i\alpha}\Psi_i$. This guarantees the stability of the lightest dark baryon.[21]

In dark QCD the dark baryons are states composed of $N$ valence dark fermions, since these form a gauge invariant object when contracted with the $\mathrm{SU}(N)_{\mathrm{DC}}$ anti-symmetric tensor. For an even value of $N$ the dark baryons are bosons, while for an odd value of $N$ they are fermions, possibly with a large spin. If the lightest dark baryon, i.e., the DM candidate, is composed of electrically charged constituents, it will have a large magnetic moment, in the same way as in the SM the neutron has a large magnetic moment, even though it is electrically neutral. If DM has a large magnetic moment this could be resolved in direct detection experiments, since it would lead to a peculiar scattering pattern (both in terms of deposited energy and in the dependence of scattering rates on chosen target nucleus, see around eq. (5.69)).

The masses $m_i$ of the dark DC fermions $\Psi_i$ are free parameters. Just as in QCD, the dark constituents can be either lighter or heavier than the dark confining scale, $\Lambda_{\mathrm{DC}}$. As mentioned above, the two limits result in bound states with qualitatively different properties.

1. If constituents are lighter than $\Lambda_{\mathrm{DC}}$, then the DM properties are determined by the non-perturbative regime of the $\mathrm{SU}(N)_{\mathrm{DC}}$: DM has a mass $M \approx \Lambda_{\mathrm{DC}}$, spatial size $1/\Lambda_{\mathrm{DC}}$, and a non-perturbative annihilation cross section $\sigma \approx 1/\Lambda^2_{\mathrm{DC}}$. In this case, the thermal relic abundance is reproduced for $M \approx 100\,\mathrm{TeV}$, i.e., saturating eq. (4.26).[22]

2. If, on the other hand, *all the dark constituents have a mass $m_i$ above* $\Lambda_{\mathrm{DC}}$, then the relevant $\mathrm{SU}(N)_{\mathrm{DC}}$ dynamics is perturbative. In this case DM is composed only of the lightest such constituents (of mass $m$). The DM in this case behaves as a perturbative bound state (similar to hadrons made of heavy $c, b$ quarks in QCD): the DM mass is $M \approx Nm$, and the annihilation cross section is set by its Bohr-like radius $1/m\alpha_{\mathrm{DC}}$.

Similar conclusions can be reached for the $\mathrm{SO}(N)_{\mathrm{DC}}$ and $\mathrm{Sp}(N)_{\mathrm{DC}}$ dark gauge groups. In the $\mathrm{SO}(N)_{\mathrm{DC}}$ models the dark baryons composed from $N$ fermions do exist and are also stable, thanks to a $\mathbb{Z}_2$ symmetry. Among others, this means that two dark baryons can now annihilate. In models with the $\mathcal{G} = \mathrm{Sp}(N)_{\mathrm{DC}}$ gauge group, on the other hand, it is possible to have stable dark meso-baryons composed out of two dark fermions (mesons and baryons are the same object for $\mathrm{Sp}(N)_{\mathrm{DC}}$).

---

[21]Like the SM baryon number, also the dark baryon number can be broken by weak anomalies. Their effects are exponentially suppressed, so the lightest dark baryon can usually be treated as effectively stable for all practical purposes.

[22]If the dark QCD phase transition is first order and/or if the model is non-minimal with light degrees of freedom that decay slowly, the expected value for $M$ can be even multi PeV [820].

| Gauge group $\mathcal{G}$ | Fermion symmetries $\mathcal{G}_F \to \mathcal{H}_F$ | Scalar symmetries |
|:---:|:---:|:---:|
| $SU(N)_{DC}$ | $SU(N_F)_L \otimes SU(N_F)_R \to SU(N_F)$ | $U(N_S)$ |
| $SO(N)_{DC}$ | $SU(N_F) \to SO(N_F)$ | $O(N_S)$ |
| $Sp(N)_{DC}$ | $SU(N_F) \to Sp(N_F)$ | $Sp(2N_S)$ |

Table 9.4: *Assuming the confining dark gauge groups listed in the first column, we show in the second column the patterns of chiral symmetry breaking induced by condensates of $N_F$ light fermions in the fundamental representation. In the third column we show the symmetry associated to $N_S$ light scalars in the fundamental. It can be broken by condensates or vacuum expectation values in different ways. See, e.g. [821] for extra details.*

### Dark glue-balls

The scenario discussed at point 2. above — a dark gauge group $\mathcal{G}$ with all fermions or scalars heavier than its confinement scale $\Lambda_{DC}$ — predicts, in addition to heavy dark baryons as DM candidates, the possible existence of dark glue-balls with mass $M_{DG} \sim$ few $\Lambda_{DC}$. These are bound states consisting of (dark) gauge bosons alone. As DM candidates, they are typically unfit because they are unstable. In addition, they can be long-lived enough that they become problematic in the cosmological context. On the other hand, their small mass makes them a good target in the context of searches of dark sector states at colliders or accelerators.

For example, the lightest glue-ball is likely the state corresponding to the gauge-invariant operator $\text{Tr}\,\mathcal{G}^2$, with quantum numbers $J^{CP} = 0^{++}$, meaning that it can be thought as a two dark-gluon state ($\mathcal{G}$ is the matrix of dark gauge vectors in the adjoint representation of $\mathcal{G}$).

In the presence of heavy fermions or scalars charged under both $\mathcal{G}$ and $\mathcal{G}_{SM}$, two dark-gluons can annihilate into SM vectors $VV$ (gluons or electro-weak). This means that the dark-glue-ball decays into $VV$ via the dimension-8 operator $\sim \alpha_D \alpha_{SM} \text{Tr}\,G^2 \text{Tr}\,V^2/M^4$, with rate

$$\Gamma_{DG} \sim \frac{\alpha_D^2 \alpha_{SM}^2 \Lambda_{DC}^9}{4\pi M^8}. \tag{9.33}$$

If $M \gtrsim 10^3 \Lambda_{DC}$ the dark glue-ball life-time exceeds one second and conflicts with Big Bang Nucleosynthesis bounds. Dark glue-balls can also decay faster, via the dimension-6 $(\text{Tr}\,G)^2|H|^2$ operator, in models where dark quarks have Yukawa couplings to the Higgs doublet $H$. In an effective field theory context below the weak scale one can also have dimension-7 $(\text{Tr}\,G)^2 f\bar{f}$ operators involving SM fermions $f$.

In the absence of any additional matter content, the dark sector has no renormalizable interactions with the SM. The lightest dark glue-ball can thus decays only gravitationally (see section 9.8), with slow rate $\Gamma \sim M^5/M_{Pl}^4$ and therefore is an acceptable DM candidate with life-time $\tau \gtrsim 10^{26}$ sec (see section 6) if $M \lesssim 100\,\text{TeV}$ [154]. Depending on the dark gauge group $\mathcal{G}$, other dark states beyond the lightest glue-ball may be metastable in view of a C-parity and/or a P-parity in the dark gauge group. These parities are discussed in general in section 10.6.3, and we here summarise their implications for glue-balls. If the dark gauge group is $\mathcal{G} = SU(N)$, the spectrum contains C-odd glue-balls that correspond to the operator $\text{Tr}\,G\{G,G\}$. The lightest of these is gravitationally stable and decays with a highly suppressed rate $\Gamma \sim M(M/M_{Pl})^8$, induced by possible non-renormalizable operators that break the dark C-parity. The $\mathcal{G} = SO(N)$ theories with $N \geq 6$, and $N$ even, contain glue-balls odd under a parity P in the dark group and corresponding to the operators $\text{Pf}\,\mathcal{G}$. The lightest of these is also gravitationally stable and decays only in the presence of non-renormalizable operators that break dark P-parity, leading to the highly suppressed decay rate $\Gamma \sim M(M/M_{Pl})^{2N-4}$. P-odd glue-balls therefore constitute acceptable gravitational DM candidates even for very large values of $M$ (for reasonably large $N$).

## QCD-like dark pions

If $N_F$ dark fermions are lighter than $\Lambda_{\rm DC}$ then the dark sector Lagrangian satisfies an approximate accidental global symmetry $\mathcal{G}_F$. For instance, in dark QCD this is $\mathcal{G}_F = {\rm SU}(N_F)_L \otimes {\rm SU}(N_F)_R$ under which the dark fermions transform as $\Psi_{L/R,i} \to [\exp(i\alpha^a_{L/R}T^a)]_{ij}\Psi_{L/R,j}$, with $T^a$ the generators of ${\rm SU}(N_F)$ in the fundamental representation and $\alpha^a_{L,R}$ free parameters (the symmetry is exact in the limit $m_i \to 0$). The dark fermion condensates break $\mathcal{G}_F$ to its subgroup $\mathcal{H}_F$ as listed in table 9.4.[23] The symmetry breaking condensates are either $\langle \Psi\bar{\Psi}\rangle \neq 0$, in the case that the dark fermions are in a complex representation, e.g., in the fundamental of $\mathcal{G} = {\rm SU}(N)_{\rm DC}$, or they can be $\langle \Psi\Psi\rangle \neq 0$, e.g., for dark fermions that are in the fundamental representation of ${\rm SO}(N)_{\rm DC}$ or ${\rm Sp}(N)_{\rm DC}$.

The spontaneous breaking of a global symmetry implies the existence of pseudo-Nambu-Goldstone bosons (pNGB) — pion-like scalar bound states lighter than $\Lambda_{\rm DC}$ — which parametrise the $\mathcal{G}_F/\mathcal{H}_F$ coset. The existence of dark pions is phenomenologically important.

First of all, even for models in which the DM candidate is a dark baryon, the first collider signatures of such models could be the production of dark pions rather than DM. Dark pions are normally lighter than dark baryons, and thus easier to produce. In general, they are also unstable, since they can decay into SM vectors through anomalies, and are thus easier to observe.

Secondly, if the dark sector Lagrangian is invariant under additional symmetries, some of the *dark pions can actually be stable and be DM candidates*. A common possibility for such an additional symmetry are the dark species numbers. These arise if dark fermions $\Psi_i$ belong to two or more irreducible representations of the SM gauge group (that is, if dark sector is composed of several different species of dark fermions, $\Psi_a$, each furnishing a different irreducible representation of the SM gauge group, while being in the same representation of ${\rm SU}(N)_{\rm DC}$). Depending on the full field content of the theory, the phase rotations acting individually on each representation $\Psi_a$ may be an accidental 'dark-flavor' symmetry. For instance, in the dark QCD Lagrangian of eq. (9.32) the dark species number is an accidental symmetry for all dark species for which the Yukawa couplings vanish, $y^{L;R}_{aj} = 0$. Dark pions composed of different species of dark fermions, $\pi_{ab} \sim \bar{\Psi}_a\Psi_b$, are then stable, at least as far as renormalizable interactions are concerned. Heavy UV physics can alter these conclusions. In many models, such flavour symmetries are explicitly broken by dimension-5 or dimension-6 operators, which then lead to too fast decays of dark pions for these to be considered viable DM candidates.

It is also possible to have accidental symmetries that are of group-theoretical nature. For example, some models exhibit a symmetry analogous to the $G$-parity in QCD, $\Psi_i \to [\exp(i\pi T^2)]_{ij}\Psi_j$, leading to a stable lightest dark-pion that can be a DM candidate [819]. This happens in models with the ${\rm SU}(N)_{\rm DC}$ gauge interaction that have $N_F = 3$ dark fermions in the triplet (real) representation of the electroweak ${\rm SU}(2)_L$ gauge group. The ${\rm SU}(N_F)_L \otimes {\rm SU}(N_F)_R \to {\rm SU}(N_F)$ spontaneous breaking of the flavor groups results in $N_F^2 - 1$ pseudo-Goldstone bosons that form a $3 \oplus 5$ representation of ${\rm SU}(2)_L$. Because of $G$-parity, the triplet pNGB is stable and has the phenomenology of the Minimal Dark Matter scalar triplet, see section 9.3.4.

If DM is a dark pion, this has other phenomenologically interesting consequences. For example, various strongly interacting theories allow for interactions involving an odd number of dark pions. such as the Wess-Zumino-Witten interaction among 5 dark pions. The resulting DM DM DM $\leftrightarrow$ DM DM scattering at nearly strong coupling realises the scenario for a sub-GeV thermal relic DM discussed in section 4.1.10 [140].

---

[23]Note that some works study effective field theories of composite dark particles assuming arbitrary $\mathcal{G}_F$ and $\mathcal{H}_F$ without discussing how such breaking patterns are achieved from microscopic dynamics of constituents. It is far from clear that such general symmetry breaking patterns can in fact be realized.

## 9.5.2   QCD-unlike strongly coupled theories

While dark QCD may be a useful benchmark model for composite DM, it by no means exhausts all the possibilities. Other possibilities include:

- Some strongly coupled gauge theories with vector-like fermions result in **confinement without chiral symmetry breaking**, corresponding to vanishing fermions condensates. This new phenomenon was proven in some supersymmetric theories. A non-supersymmetric example might be SU($N$) with 3 fermions in the adjoint [822]. In these models, dark-colored fermions much lighter than the confinement scale must form composite fermions much lighter than the confinement scale, $M \ll \Lambda_{\rm DC}$, as demanded by 't Hooft anomaly matching conditions [822]. Such composite dark baryons might be accidentally meta-stable DM candidates, and would behave as elementary particles up to corrections of order $M/\Lambda_{\rm DC}$ [822], differentiating them from QCD-like baryons such as the proton.

- Dark sector with gauge interactions but with matter content composed of **massless chiral fermions** is another possibility [823]. This is theoretically appealing, since all scale(s) $\Lambda_{\rm DC}$ are generated by dimensional transmutation. DM can similarly be dark protons or dark pions. These models tend to employ multiple gauge groups in order to avoid gauge anomalies, for example mimicking a chiral family of the SM by postulating extra U(1) ⊗ SU(2) ⊗ SU(3) dark groups, or partially employing the factors of the SM gauge group. Multiple gauge groups allow to get chirality from a weakly coupled factor, avoiding the poorly known dynamics of strongly coupled chiral theories.[24] The more complicated chiral models can lead to dark pions as the DM, with the DM decay rates more suppressed than in the simple vector-like models.

- The matter content in the dark sector could be composed of **dark scalars** instead of dark fermions, or one could have both dark scalars and dark fermions. This more general field content then allows to build explicit models where both DM and the Higgs are composite [821]. However, less is known about strong gauge dynamics of scalars. The last column in table 9.4 shows, for each gauge group class, the global flavor symmetries for models containing $N_S$ scalars, assuming these are unbroken. In explicit models the flavor symmetries can be broken, with the symmetry breaking pattern being model-dependent.

- One can consider theories where, instead of gauge interactions becoming non-perturbative, the strong coupling is either a **Yukawa interaction** or **scalar quartics**.

## 9.5.3   Weakly-coupled bound states: dark atoms and mirror DM

Even if the interactions in the dark sector are perturbative, dark matter particles can still form (weakly) bound dark states. These can be unstable, for instance if they are formed from DM and its antiparticle [133], akin to quarkonia in QCD, or can be stable, if they are formed from two different species of DM particles. The formation of unstable bounds states, whose constituents eventually annihilate, can be an important effect in the early Universe and can change significantly the predicted relic abundance. They were discussed, together with Sommerfeld corrections, in section 4.1.5. Stable bound states, instead, are the so called **dark atoms** [824] and we focus on them in the following.

---

[24]In chiral theories, the QCD-like scalar condensates of two fermions $\langle \bar{\Psi}\Psi \rangle$ are forbidden by gauge invariance (because, if $\bar{\Psi}\Psi$ were allowed, one could add mass terms to fermions, and the theory would not be chiral). So strongly-coupled chiral gauge theories could either form no condensates, or form vector condensates $\langle \bar{\Psi}\gamma_\mu\Psi \rangle \neq 0$. In such a case the Lorentz group would be broken spontaneously in the dark sector — a possibility disfavoured by the smallness of the cosmological constant and by bounds on Lorentz symmetry breaking.

The simplest example of a dark sector in which dark atoms form is a model with two species of dark fermions, a dark proton and a dark electron, that carry opposite charges under a hidden $U(1)_D$ gauge symmetry,

$$\mathscr{L}_D = \bar{\Psi}_{Dp}\left(i\slashed{D} + m_{Dp}\right)\Psi_{Dp} + \bar{\Psi}_{De}\left(i\slashed{D} + m_{De}\right)\Psi_{De}, \tag{9.34}$$

with $D_\mu = \partial_\mu + ig_D Q_D A_\mu^D$, and $Q_D = \pm 1$ for dark proton and electron, respectively. This model has three parameters, the dark gauge coupling constant, $g_D$, and the masses of dark proton and electron, $m_{Dp}$, $m_{De}$ (without loss of generality one can define the dark proton such that, $m_{Dp} > m_{De}$). The formation of dark atoms is one of those cases in which the existence of an asymmetry in the dark sector is typically required, i.e. a relic abundance of dark protons and dark electrons without dark anti-protons and anti-electrons, analogous to what occurs in the SM sector.

Due to the presence of dark atoms, the cosmological evolution is richer than in the case of a single elementary DM species. In general, the dark sector temperature $T'$ will differ from the temperature $T$ in the visible sector. Furthermore, the ratio $T'/T$ can change during the evolution of the Universe, following different entropy injections in the two sectors. If the dark photon is massless, its two degrees of freedom account for an extra contribution to the relativistic energy density $\Delta N_{eff} = 8/7\,(11/4)^{4/3}\,(T'/T)^4$ (see eq. (C.27) in App. C.3). The current rather loose bound $\Delta N_{eff} < 0.33$ at $2\sigma$ [3] implies the not very stringent constraint $T' < 0.55\,T$. In dark atom models the dark sector interactions can decouple much later than in the conventional cold DM models with elementary DM particles. This has implications for structure formation: for instance, the proto-halo formation can be suppressed for proto-halos with masses below $\sim 10^3 - 10^6 M_\odot$. An important aspect of dark atom models is that now DM is a mixture of neutral and ionised components. The ionized fraction is phenomenologically important since it interacts through a long range force: bounds on self-interactions of DM (see section 3.3.3) are an important constraint. The self-interactions lead to energy dissipation (the same as electromagnetic interactions do for visible baryons), which then in general leads to denser, cuspier DM profiles [824]. Another potential effect of dissipation is that (a subdominant component of) atomic DM can form many intermediate-mass black holes that act as seeds for the otherwise puzzling existence of early massive BHs [824].

Assuming interactions with the visible sector, these lead to scattering in direct detection experiments as for usual cold DM. The difference with the elementary DM is that the scattering of dark atoms can result in outgoing dark atoms in an excited state: if these are short-lived they can result in additional signatures in the detectors, if the decays to visible sector particle are frequent enough.

Within the dark atom hypothesis it is quite natural to entertain also the existence of dark molecules, dark compounds, and an entire *dark chemistry*. A few recent works have started exploring the implications of these concepts, especially in star formation and the early Universe evolution [824].

The special case in which the free parameters of the dark atoms ($g_D$, $m_{Dp}$ and $m_{De}$) coincide with the values in the visible sector is called *mirror world* [771]. In this setup, the entire SM has a mirror copy, including quarks and leptons and force mediators. In particular, the baryons of the mirror sector are, like the visible ones, stable particles; they interact with mirror photons but they are dark in terms of the ordinary photons. Hence they could constitute a DM candidate known as the **mirror DM**.

The two sectors communicate via gravity but also via the mixing of the neutral SM and mirror particles: photons, neutrons, neutrinos. The latter two mixings can give rise to co-baryogenesis/leptogenesis, inducing comparable baryon asymmetries and hence $\Omega_{DM} \sim \Omega_b$. The mirror symmetry cannot however be exact and, in particular, the mirror sector needs to have a lower temperature in order to agree with cosmology, including cold mirror DM, as discussed above. This can be realized if the two systems are born with different temperatures at reheating.

Besides providing mirror DM, the mirror world theory also makes several specific cosmological and astrophysical predictions. Mirror BBN would lead to a higher fraction of mirror He, so that mirror stars would be bigger, possibly seeding the observed ultra-heavy black holes. The bullet cluster bound (see section 1.2.1) can be satisfied if at least half of the mirror sector DM is in dark stars, rather than in gas.

If the baryon asymmetries have opposite sign, $\bar{n} \leftrightarrow n'$ oscillations would allow appearance of $\bar{n}$ out of the mirror sector (for example in neutron stars), possibly allowing to explain the anti-matter events claimed by AMS (section 8.2.10) and providing a connection with the neutron decay anomaly (section 8.3.3).

Before concluding this section, we mention the proposal of **O-helium** [825], a model that is sometimes referred to as 'dark atom' but is actually a hybrid of the strongly-coupled DM described in section 9.5.1 and the atomic physics discussed here. In this model, a techni-color-like or a QCD-like theory leaves stable, massive (typically $M \sim$ TeV), doubly charged $O^{--}$ particles that then bind to $He^{++}$ nuclei just after nucleosynthesis. The exotic $O^{--}He^{++}$ 'atom' may constitute the DM and may be searched for with the techniques discussed in section 5.1.2.

## 9.6  Dark magnetic monopoles

Quantum field theory predicts a variety of topological defects that can arise when a gauge symmetry group $\mathcal{G}$ is spontaneously broken to its sub-group $\mathcal{H}$. An example are magnetic monopoles [826], particle-like configurations of gauge and scalar fields concentrated around a point in space, which are made stable by topology when the second homotopy group $\pi_2(\mathcal{G}/\mathcal{H}) \neq 0$ is non-trivial [827]. Loosely speaking, this means that there exists a mapping of a sphere in physical space (the boundary of 3-dimensional space at infinity) into a sphere in the coset $\mathcal{G}/\mathcal{H}$, which cannot be continuously shrunk to a single point in $\mathcal{G}/\mathcal{H}$.[25] Since it is impossible to unravel a topologically nontrivial configuration in $\mathcal{G}/\mathcal{H}$, the magnetic monopole is stable, which is a good start to be considered as a DM candidate.

As anticipated in section 9.1.4, the SM electro-weak phase transition does not lead to monopoles, since the associated second homotopy group is trivial, $\pi_2(\mathrm{SU}(2)) = 0$. We then need to consider new-physics models [167]. Let us focus on the simplest possibility, $\mathcal{H} = \mathrm{U}(1)$. This $\mathrm{U}(1)$ cannot be the usual electro-magnetism, since electro-magnetic monopoles are subject to strong experimental bounds — DM cannot be a magnetic monopole of the usual electro-magnetism. The simplest example of a theory with a dark sector magnetic monopole is thus a dark $\mathcal{G} = \mathrm{SU}(2)$ gauge group with a scalar $S$ in the adjoint representation. The dark $\mathrm{SU}(2)$ is spontaneously broken to its $\mathrm{U}(1)$ subgroup once $S$ obtains a vacuum expectation value, $\langle S \rangle = w \, \mathrm{diag}(1,-1)/2$ (Khoze and Ro (2014) in [167]). At the perturbative level, the spectrum contains a massive Higgs-like scalar (neutral under the dark $\mathrm{U}(1)$), a massless dark photon $A'$,[26] and massive charged vectors $V^{\pm}$ (charged under dark $\mathrm{U}(1)$). The charged vectors have mass $M_V = gw$, carry a charge $g$ under the dark $\mathrm{U}(1)$, are stable and are thus good DM candidates ($g$ is the dark gauge coupling). The non-perturbative spectrum contains magnetic monopoles which have a mass

---

[25]More concretely, consider a scalar field $\varphi$ in some representation of group $\mathcal{G}$. Its expectation value results in $\mathcal{G} \to \mathcal{H}$ spontaneous symmetry breaking. A magnetic monopole corresponds to a topologically stable configuration of $\varphi$ of finite energy. For this to be the case the scalar potential needs to vanish far from the origin, $\lim_{|\boldsymbol{x}|\to\infty} V(\varphi(\boldsymbol{x})) = 0$, while the value of $\lim_{|\boldsymbol{x}|\to\infty} \varphi(\boldsymbol{x}) \equiv \varphi(\hat{\boldsymbol{x}})$ is nonzero, and is different in different directions $\hat{\boldsymbol{x}}$. With a particular $\varphi(\boldsymbol{x})$ configuration there is also an associated configuration of unbroken gauge fields, which then define the magnetic monopole. Since $\varphi$ does not change under $H$, the values $\varphi(\hat{\boldsymbol{x}})$ live in the $\mathcal{G}/\mathcal{H}$ coset, and give the mapping of the space boundary, the two-dimensional sphere $S_2$, into the $\mathcal{G}/\mathcal{H}$ coset. If this mapping is topologically nontrivial, i.e., there is no continuous deformation that can make $\varphi(\hat{\boldsymbol{x}})$ vanish while keeping $V(\varphi(\hat{x})) = 0$, the configuration is stable. This is possible only, if $\pi_2(\mathcal{G}/\mathcal{H}) \neq 0$. For example, for any simply connected Lie group $\mathcal{G}$ broken to $\mathcal{H}$ one has $\pi_2(\mathcal{G}/\mathcal{H}) = \pi_1(\mathcal{H})$, so that there are monopoles, if the first homotopy group of the unbroken subgroup is nontrivial, i.e., there are nontrivial mappings of a circle into $\mathcal{H}$. This is, for instance, the case for $\mathcal{H} = \mathrm{U}(1)$, for which $\pi_2(\mathcal{G}/\mathrm{U}(1)) = \pi_1(\mathrm{U}(1)) = \mathbb{Z}$, i.e., there is a discrete number of monopoles that carry integer magnetic charges $k$.

[26]Massless vectors result in extra dark radiation in cosmology. Current bounds are satisfied, if the dark sector decouples from the SM early enough (say, before the QCD phase transition), and is thus significantly colder than the SM plasma at later stages of the cosmological evolution.

$M = 4\pi w/g$ (up to corrections from possible non-vanishing scalar quartics). The magnetic monopoles carry a magnetic charge $g_{\rm mag} = 4\pi/g$ under the dark U(1), are stable, and are also viable DM candidates.

The monopoles are produced during the $\mathcal{G} \to \mathcal{H}$ phase transition which occurs at temperature $T \sim M_V$. The production rate depends on the nature of the phase transitions, see section 4.5.1 for details. In particular, for this simplest model under consideration, the dark monopoles can constitute an $\mathcal{O}(1)$ fraction of DM relic abundance if the phase transition is second order, and in that case they have roughly TeV to PeV mass. For (strongly) first order phase transition, instead, the DM relic abundance is entirely dominated by the heavy dark vectors produced through perturbative thermal freeze-out and dark monopoles are only a very small fraction of DM, below $10^{-10}$.

These conclusions likely apply quite generally, but may differ in details for other models. One possible difference concerns, for example, monopole annihilations. In general, monopoles can annihilate with anti-monopoles of opposite magnetic charge. In the model we considered this is the only annihilation channel. Different models allow, however, also for the annihilation of two monopoles of the same charge (e.g. Bai, Korwar and Orlofsky (2020) in [167]). This happens if the unbroken group is $\mathcal{H} = {\rm SO}(n)$ with $n \geq 3$ such that $\pi_2(\mathcal{G}/{\rm SO}(n)) = \pi_1({\rm SO}(n)) = \mathbb{Z}_2$, meaning that there is only one monopole, corresponding to the element $-1$ of $\mathbb{Z}_2$. This distinction is quite analogous to what happens for the point-like particles. Monopole annihilations are one possible avenue for their indirect detection, see section 6. If the field content of the model is such that there are fields charged under both the SM and dark gauge groups, this induces a kinetic mixing between the dark photon and the photon. In this case magnetic monopoles become a decaying DM candidate, slowly decaying into SM particles and with the phenomenology discussed in section 6.

Since in the above models of dark monopoles DM particles interact with the massless U(1) 'dark photon', either electrically or magnetically, this leads to observable consequences. The dark sector plasma in the Early Universe now forms an interacting fluid rather than a free-streaming one. The figure of merit is the transfer cross section (the cross section weighted by the fractional longitudinal momentum transfer[27]), which for scattering of two heavy vector DM particles is given by (see, e.g., Feng et al. (2009) and Tulin et al. (2013) in [119])

$$\sigma_{\rm tran} = \frac{g^4}{\pi M_V^2 v^4}\ell, \tag{9.35}$$

where $\ell \sim \mathcal{O}({\rm few}\,10)$ is a logarithmic IR enhancement, cut-off by the small thermal mass acquired by dark photons propagating in a dark sector plasma. Assuming that the DM relic abundance is dominated by the heavy dark vectors, the bullet cluster and other observations bound (see eq. (1.5)) $\sigma_{\rm tran}/M_V \lesssim {\rm cm}^2/{\rm g} \sim 4580/\,{\rm GeV}^3$ at $v \sim 1000\,{\rm km/s}$, implying $M_V \gtrsim g^{4/3}\,300\,{\rm GeV}$. When such bound is nearly saturated, self-interacting DM can marginally improve the potential observational problems, such as core-vs-cusp, too-big-to-fail, etc, discussed in section 8.5. A weaker bound is obtained by imposing that DM free-streams during structure formation at $T \sim T_{\rm eq} \sim$ eV, giving $\sigma_T T^4/M \lesssim H$ with $\sigma_T = g^4/6\pi M^2$. The same results hold for when dark monopoles dominate, but with the replacements $g^2/4\pi \to 4\pi/g^2$, $M_V \to M$. When the fractions of elementary DM and dark monopole DM are comparable one needs to appropriately interpolate between the two versions of constraints.

Finally, we remark that the occurrence of both particles and monopoles is not a specific complication of the sample models we considered. Indeed, gauge theories are conjectured to have two equivalent descriptions [826]. The first is in terms of $\mathcal{H}$ gauge fields, with elementary charged particles and with magnetic monopoles as solitonic solutions. The second is a description in terms of a dual gauge group $\tilde{\mathcal{H}}$, where magnetic monopoles are the elementary particles and charged particles are the solitonic solutions. This implies that genuinely new classes of DM models are only the ones where both magnetic monopoles and elementary charged particles are present; a theory with only magnetic monopoles and gauge fields in

---

[27]The cross section between two DM particles is instead infinite, due to the long-range of Coulomb interaction.

the spectrum can be traded for a dual description of elementary particles charged under the dual gauge group.

## 9.7 Asymmetric DM

For DM that is not its own anti-particle, so that it carries a conserved dark number $D$, the observed DM relic abundance could be due to a small asymmetry in the abundances of DM and $\overline{\text{DM}}$ in the early Universe. Even if the symmetric populations largely annihilate away during the cosmological evolution, the asymmetric part remains. This idea of *asymmetric DM* [165] is very similar to the way baryons are believed to be generated in the early Universe through a process of baryogenesis. The mechanism of baryogenesis is not known experimentally, and many theories involving beyond the SM physics exists: the same is true for asymmetric DM.

An especially interesting possibility is that baryogenesis and the generation of asymmetric DM are part of the same mechanism, the so called *co-genesis* (see also the discussion in section 4.4). The main idea is that in the early Universe there are interactions that can convert SM particles to dark sector particles, so that the $B - L - D$ quantum number is conserved, but not $B - L$ and $D$ separately ($B + L$ is broken by electroweak sphalerons). If an asymmetry in $B - L - D$ is generated, this is then shared between the dark and visible sectors. Once these interactions decouple, $B - L$ and $D$ separately become conserved, and the asymmetries in $B - L$ and $D$ are frozen, with their values linked through its previous shared thermal history.

As a concrete example let us consider a modified version of baryogenesis via leptogenesis [164]. This can be a model of asymmetric DM, if the heavy right-handed neutrinos, $N_i$, have Yukawa couplings to both the SM leptons $L$ and Higgs $H$, as well as to dark sector fermions $L'$ and dark scalars $H'$:

$$\mathscr{L} = \mathscr{L}_{\text{SM}} + \bar{N}_i i \slashed{\partial} N_i - \left( \frac{M_{N_i}}{2} N_i^2 + y_i N_i H L + y_i' N_i H' L' + \text{h.c.} \right). \tag{9.36}$$

The Yukawa couplings in the visible sector generate neutrino masses $m_\nu \sim y^2 v^2 / M_N$, and thus for large Yukawa couplings, $y \sim 1$, the right-handed neutrinos need to be very heavy, $m_N \sim 10^{15}$ GeV. There are different options for what dark sector particles can be, for instance, both $L'$ and $H'$ could be SM singlets, or they could be in some non-singlet representation of the SM gauge group, in which case the SM gauge interactions already provide an efficient annihilation mechanism. For concreteness let us assume that $L'$ is the DM, and assign it a dark number $D = 1$, and $B - L = 0$. The interactions in eq. (9.36) conserve $B - L - D$ (with $N_i$ and $L$ carrying $B - L - D = \pm 1$, respectively). In particular, in the early Universe the scatterings $HL \leftrightarrow N_i \leftrightarrow L'H'$ convert visible to dark states and vice versa. If some mechanism creates an asymmetry in $B - L - D$ this is then shared between visible and dark sectors. Once the temperature drops well below the $N_i$ massses, the scatterings are no longer effective, and the asymmetries in $B - L$ and $D$ become frozen (but linked numerically).

The opposite limit is that the $B - L$ and $D$ asymmetries are generated after the two sectors are already out of chemical equilibrium. This is, for instance, the case if the asymmetries are generated by the out-of equilibrium decays of the lightest right-handed neutrino $N_1$, which can decay quite slowly, if the Yukawa couplings are small, with rate $\Gamma_{N_1} \approx [y_1^2 + \mathcal{O}(1)y'^2]M_1/8\pi$. The interferences of tree and one loop diagrams result in CP asymmetries for individual decay channels,

$$\varepsilon \equiv \frac{\Gamma(N_1 \to LH) - \Gamma(N_1 \to \bar{L}H^*)}{\Gamma(N_1 \to LH) + \Gamma(N_1 \to \bar{L}H^*)} \sim \frac{1}{4\pi} \frac{M_{N_1}}{M_{N_{2,3}}} \text{Im } y_{2,3}^2, \tag{9.37a}$$

$$\varepsilon' \equiv \frac{\Gamma(N_1 \to L'H') - \Gamma(N_1 \to \bar{L}'H'^*)}{\Gamma(N_1 \to L'H') + \Gamma(N_1 \to \bar{L}'H'^*)} \sim \frac{1}{4\pi} \frac{M_{N_1}}{M_{N_{2,3}}} \text{Im } y_{2,3}'^2. \tag{9.37b}$$

Note that the asymmetries vanish in the limit of all couplings being real, since in that case there is no CP violation. The $\varepsilon$ asymmetry results in a lepton asymmetry in the visible sector (leptogenesis), which then gets converted to baryon asymmetry via electroweak sphalerons (baryogenesis via leptogenesis). The $\varepsilon'$ asymmetry leads to an asymmetry in the $L'$ and $H'$ populations in the dark sector. Once the symmetric populations annihilate away this then results in an asymmetric population of DM, either from $L'$ or $H'$ being the DM, or from their further decays to DM. Eq.s (9.37) also show that the decay rates and asymmetries in the SM and dark sector are in general different, depending on what the unknown values of the $y_i$ and $y_i'$ Yukawa couplings are. However, if these are comparable, and in addition the DM mass is comparable to proton mass, then the DM and baryon relic abundances are predicted to be similar, which one can take as an explanation of the fact that the observed values are comparable, $\Omega_{\mathrm{DM}} \approx 5\Omega_{\mathrm{b}}$ (a fly in the ointment being that in this case it is also very easy to get vastly different abundances). Note also, that the proton mass is given by $\Lambda_{\mathrm{QCD}}$, and is thus exponentially sensitive to the quark content of the SM (since this affects the RG running of $\alpha_{\mathrm{s}}$) and to the value of the strong coupling $\alpha_{\mathrm{s}}$ at the UV scale (since this sets the initial condition for the RG running). Having a DM mass comparable to the proton mass is therefore natural only in special theories, such as a dark sector that is a near-copy of the SM [165].

The simple example above illustrates the salient features of many cogenesis models. A successful model of baryogenesis needs to satisfy the three Sakharov conditions [163]: *i)* departure from thermal equilibrium, *ii)* sufficient C and CP violation, *iii)* violation of baryon number. In the above 'baryogenesis via leptogenesis' example these are: *i)* the out-of-equilibrium decays of $N_1$, *ii)* the CP violating couplings of $N_1$, and *iii)* the $B+L$ violating interactions via electroweak sphalerons. However, for co-genesis models the third condition is not really needed: if DM carries baryon number, all the processes can be baryon number conserving, so that one ends up with equal and opposite baryon number densities in visible and dark sectors (this would, for instance, be the case if $N_1$ had decays of the form $N_1 \to uddX_{\mathrm{DM}}$ with $u, d$ the SM quarks). Since the cogenesis models introduce mostly new states or states only feebly coupled to the SM, the parameters of the models are poorly constrained by observations. This means that there is in general significant freedom in the predicted value of $\Omega_{\mathrm{DM}}$.

This is especially the case for the so called *darkogenesis* variants of the asymmetric DM models, in which the asymmetry in $D$ is first generated in the dark sector, and then transferred to the $B - L$ asymmetry in the visible sector. In this case, there is significant freedom of choice for both the generation and transfer mechanisms, as well as for their parameters. In general, any model of baryogenesis can be turned into a model of darkogenesis: decays, oscillations, a first order phase transitions in the dark sector, scalar condensates, etc.

## 9.8 Gravitational Dark Matter

So far, we only observed gravitational interactions of DM. One can thereby consider theories where DM is a particle (a singlet scalar $S$, a singlet fermion $N$, or a massive vector $V$) endowed with only gravitational interactions [154]. The graviton interactions with matter are theoretically well known: expanding $g_{\mu\nu} = \eta_{\mu\nu} + 2h_{\mu\nu}/\bar{M}_{\mathrm{Pl}}$, the interaction of a single graviton is $\mathscr{L}_{\mathrm{int}} = h_{\mu\nu}T^{\mu\nu}/\bar{M}_{\mathrm{Pl}}$, where $T_{\mu\nu}$ is the energy-momentum tensor whose form depends on whether DM is a scalar, a fermion or a vector. At sub-Planckian energies graviton exchanges result in power suppressed interactions, with the effective coupling given by $g_{\mathrm{grav}} \sim E/M_{\mathrm{Pl}}$. For scalar DM the gravitational interactions also contain an additional non-minimal coupling, $\xi_S R S^2/2$, where $R$ is the Ricci scalar, and $\xi_S$ is a dimensionless coupling. Below, we will mostly be able to ignore the effects of this non-minimal coupling, while it needs to be taken into account for more detailed results.

In the early Universe, there are two main mechanisms for production of particles with only gravitational interactions: either through freeze-in during the thermal phase of cosmology (section 4.2.2), or

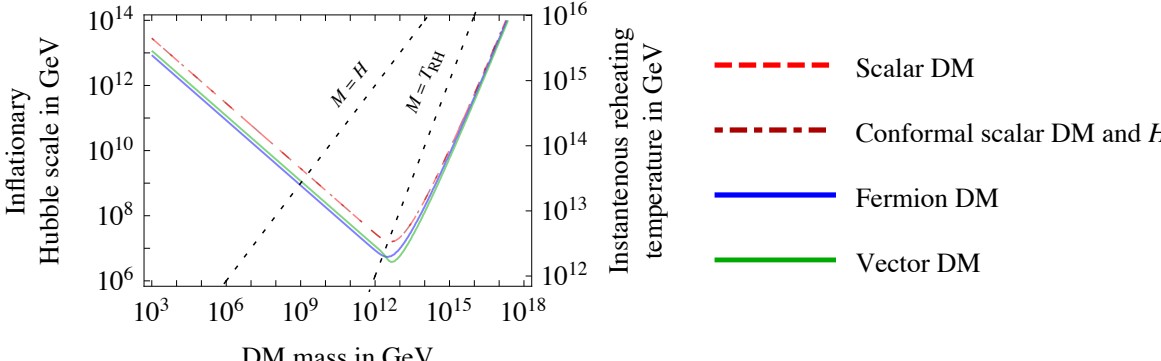

Figure 9.8: *Curves along which freeze-in thermal production of* **DM with gravitational couplings** *reproduces the DM abundance. The dashed black lines indicate at which values the DM mass M equals either the reheating temperature $T_{\mathrm{RH}}$ or the inflationary Hubble constant.*

already earlier, during inflation (section 4.3).

- **Freeze-in**. The amplitude for the SM SM $\to$ DM DM scatterings mediated by the $s$-wave graviton exchange is, at energies much above the SM particle masses, given by $\mathscr{A} = -iT_{\mu\nu}T^{\mu\nu}/\bar{M}_{\mathrm{Pl}}^2 s$, where $s$ is the center of mass energy squared. The resulting predictions for the scattering cross sections, including $\mathcal{O}(1)$ factors, can be found in [154], while here we focus on the simple scaling estimates. The space-time density of scatterings that produce DM from the SM plasma is $\gamma \sim T^8 e^{-2M/T}/M_{\mathrm{Pl}}^4$ (with an additional $T/M$ $p$-wave suppression in the case of fermionic DM, which we ignore in the estimates below). The resulting DM number abundance is

$$Y_{\mathrm{DM}} \approx \left.\frac{\gamma}{Hs}\right|_{T=T_{\mathrm{RH}}} \sim e^{-2M/T_{\mathrm{RH}}} \left(\frac{T_{\mathrm{RH}}}{M_{\mathrm{Pl}}}\right)^3, \tag{9.38}$$

  where the reheating temperature is given by $T_{\mathrm{RH}} \sim \sqrt{M_{\mathrm{Pl}}H_{\mathrm{infl}}}$, if the reheating after the end of inflation is instantaneous, while $T_{\mathrm{RH}} \sim \sqrt{M_{\mathrm{Pl}}\Gamma_{\mathrm{infl}}}$ otherwise. For any given $T_{\mathrm{RH}}$ the DM mass abundance, $\Omega_{\mathrm{DM}} \sim Y_{\mathrm{DM}}M/T_0$, is maximised for DM mass that is comparable to the reheat temperature, $M \sim T_{\mathrm{RH}}$, since this maximises the gravitational coupling while avoiding the exponential suppression in eq. (9.38). The resulting $\Omega_{\mathrm{DM}}^{\mathrm{max}} \sim M^4/M_{\mathrm{Pl}}^3 T_0$ matches the observed DM density for $T_{\mathrm{RH}} \approx M \sim 10^{12}\,\mathrm{GeV}$. This is the minimal viable reheating temperature for gravitational DM; for lower $T_{\mathrm{RH}}$ the observed DM density cannot be reproduced. For larger $T_{\mathrm{RH}}$ the observed DM density is matched for two values of DM mass, $M_{\mathrm{heavier}} \approx \mathrm{few} \times T_{\mathrm{RH}}$, and $M_{\mathrm{lighter}} \approx (10^{12}\,\mathrm{GeV})^4/T_{\mathrm{RH}}^3$, see fig. 9.8.

- **Inflation**. As we saw above, the freeze-in production of gravitational DM is dominated by the highest temperatures after the end of inflation. It is possible that an even larger effect arises during the preceding inflationary phase. As discussed in section 4.3, this includes two possibly dominant contributions that depend on the particular (unknown) model of inflation, and occur in the final phases of the inflationary epoch: the decay of the inflaton $\phi$ after the end of inflation (DM production is kinematically blocked if $m_\phi < 2M$), and the inflaton oscillations toward the end of inflation (kinematically blocked if $m_\phi \lesssim M$). The production of DM during the period of inflation itself, on the other hand, is less model-dependent, see section 4.3.2. The end result, at least for $M \sim H_{\mathrm{infl}}$, is DM with roughly thermal distribution corresponding to $T \approx H_{\mathrm{infl}}/2\pi$, and

thereby a DM abundance $n \sim M^3 e^{-M/H_{\text{infl}}}$, but the details are rather complicated. Importantly, the inflationary contribution to DM relic density can lead to problematic iso-curvature density fluctuations. However, in the case of instantaneous reheating, the inflationary contribution is negligible compared to the freeze-in contribution, and can thus safely be ignored.

From a theoretical point of view, the hypothesis that the DM particle only has gravitational interactions is quite intriguing.[28] A priori, there is no good reason to forbid interactions such as the scalar DM quartic $|H|^2|S|^2$, coupling $S$ to the Higgs boson, or a Yukawa interaction $LHN$ in the case of fermion DM $N$, or a mixing of a vector DM with the SM hypercharge gauge field, $B_\mu$. Furthermore, if the mass of the vector DM is due to a Higgs mechanism, i.e., due the vacuum expectation value of a new dark scalar, $\varphi_{\text{dark}}$, the vector DM would also inherit any non-gravitational couplings of the dark scalar (in particular, there would be interactions between $\varphi_{\text{dark}}$ and $V$). The gravitational DM hypothesis posits that all these, otherwise completely allowed renormalizable interactions, are for some reason highly suppressed to a sub-gravitational level. Furthermore, it is not enough to forbid renormalizable interactions. Non-renormalizable Planck-suppressed operators that involve DM and SM particles, such as $S^2\bar{q}q/M_{\text{Pl}}$, naively give contributions that are comparable to the graviton mediated scatterings. Such operators are unavoidably generated by the quantum effects at the loop level. While they are loop suppressed, they are also logarithmically enhanced due to renormalisation group running from Planck scale to the energy scale at which scattering occurs, and can thus compete with gravitational interactions. In general, we can also expect that such operators are generated at tree level at the Planck scale by the still poorly understood quantum gravity physics.

This does not mean that it is not possible to forbid the non-gravitational interactions via a symmetry. A particularly simple example is a dark sector that consists of just the gauge fields of a non-abelian 'dark color' group $\mathcal{G}$ [154], without any additional matter content. This gives rise to gravitationally-interacting (meta-)stable dark glue-balls, as discussed in section 9.5.1. According to the conjectured AdS/CFT correspondence some of these theories admit a dual description: special strong gauge dynamics with slowly running gauge coupling is equivalent to gravity in a warped extra dimension, and glue-balls are equivalent to Kaluza-Klein excitations of gravitons.

This suggests that the lightest Kaluza-Klein excitation of a graviton propagating in generic extra dimensions (not necessarily warped) can be one more candidate for gravitational DM. It can be stable for special geometries, such as a flat segment with an orbifold reflection symmetry, with the SM fields confined on a three dimensional boundary (see, e.g., Garny et al. (2017) in [154]). For further discussion of other realizations of DM in extra-dimensional set-ups see section 10.3 for connections with string theory, and section 10.1.3 for connections with the hierarchy problem. The supersymmetric gravitino, to be discussed in section 10.1.2, can also behave as a gravitational DM, in a particular limit.

DM particles with only gravitational interactions are very hard to search for directly (sometimes also referred as the *nightmare scenario*). Two exceptions arise in corners of the parameter space: *i)* gravitational DM with Planck-scale mass, which can be searched for in the laboratory using an array of mechanical sensors, see section 5.5.9; *ii)* 'large' extra dimensions with low quantum gravity scale comparable to the energy reached by colliders would allow, for instance, to produce KK excitations of gravitons and extra dimensional black holes. In addition, the Pao collaboration derived constraints for the case of gravitational DM particles being able to decay into the SM particle via the non-perturbative effects due to a dark gauge group [154].

---

[28]Historically, one of the first proposals are *WIMPzillas*, introduced in section 4.3.5.

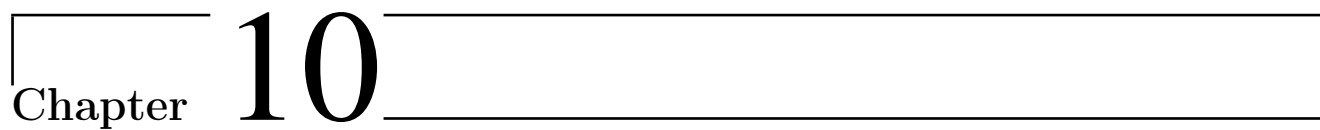

# Chapter 10

# Theoretical frameworks and Dark Matter

In this chapter we discuss bigger theoretical frameworks motivated by other considerations, that also lead to DM candidates as a side result: theories motivated by Higgs naturalness (section 10.1, including supersymmetry in section 10.1.2, composite Higgs in 10.1.4, extra dimensions in 10.1.3, ...), string theory (section 10.3), cosmology (section 10.2), the QCD $\theta$ problem and its possible axion solution (section 10.4), and macroscopic objects (section 10.5). Finally, in section 10.6 we summarise possible theoretical reasons for the stability of DM.

## 10.1 Dark Matter and Higgs naturalness

### 10.1.1 The Higgs mass hierarchy problem

The smallness of the SM Higgs mass $M_h = 125.25 \pm 0.17$ GeV [5] compared to the Planck mass, $M_h \ll M_{\rm Pl}$, (or other high scales, see below) introduces a puzzle. A calculation of one-loop quantum corrections to the Higgs mass in the SM using an explicit cutoff $\Lambda$ reveals quadratically divergent corrections, $\delta M_h^2 \sim g^2 \Lambda^2$, where $g$ denotes the various Higgs couplings entering the loop diagrams, such as the top Yukawa coupling and the weak gauge couplings. While no physical effect is associated with such corrections (that is, all physical quantities are $\Lambda$ independent, and the quadratically divergent correction itself even vanishes in dimensional regularization), the quadratic divergence signals a potential problem.

If the SM is augmented by new physics at a higher scale, such as a new heavy particle with mass $M_{\rm NP} \gg M_h$ and couplings $g$ to the Higgs, the Higgs squared mass $M_h^2$ receives regularization-independent corrections of order $\delta M_h^2 \sim g^2 M_{\rm NP}^2 \ln(E/M_{\rm NP})$, with coefficients given by the $\beta$-functions of the theory. Such effects are in principle measurable, and thereby physical.

The various contributions to $M_h^2$ can sum up to give the small physical Higgs mass, even though individual terms are much larger, of order $\mathcal{O}(M_{\rm NP}^2)$. Presumably there is at least one extra physical scale in the theory: the Planck scale (around which the non-renormalizable Einstein gravity gets replaced by a more complete theory) and possibly many others (such as the masses of extra vectors required for gauge coupling unification). In view of this, it is surprising that $M_h$ can be so small with respect to any of these new physics scales or even the Planck mass. This is the so called *hierarchy problem*.

Before the start of the Large Hadron Collider (LHC) many theorists expected special types of new physics to exist around the weak scale, such that the Higgs mass would be protected from large quantum corrections, allowing the Higgs mass to be naturally small, $\delta M_h \lesssim M_h$. The existence of Dark Matter reinforced these expectations, since thermal relics of new stable particles with electroweak-scale masses and electroweak interactions can be a viable DM candidate, with the correct cosmological abundance (see section 4.1).

Devising theoretical frameworks where new physics at the weak scale provides a natural alternative to the SM has been one of the main topics of theoretical particle physics (though, there have also been proposals that do not involve new physics at the weak scale, see section 10.1.6). The key motivation for these attempts was their imminent testability at the LHC (point 6 on page 310), which often guided the particular choices for the values of the parameters. After that the LHC searches yielded negative results, the research activity in this area declined significantly. Therefore, we will briefly highlight the main lessons and implications for Dark Matter, rather than provide a full review. A general theme is that the new weakly-interacting particles around the weak scale fit well with DM being a thermal relic.

## 10.1.2 Dark Matter and supersymmetry

Among the concrete theories of natural new physics, weak scale supersymmetry (SUSY) has long been considered by many to be the most plausible possibility. SUSY has several independent attractive features which go a long way toward explaining why this solution to the hierarchy problem used to be so popular. Below, we collect some of the main results (for a more detailed discussion see [1, 102, 828]).

In what way the Poincaré space-time symmetry could be extended is an interesting theoretical issue, which for a while seemed to be settled by the Coleman-Mandula theorem [829]. This theorem states that unless the new symmetries are internal (commute with Poincaré transformations) such theories lead to extra conserved quantities (in addition to momentum and angular momentum), forbidding scatterings among particles. Supersymmetry is the only extension that bypasses this no-go theorem by having $N$ anti-commuting parameters $\theta_\alpha$ and thereby unavoidably linking bosons to fermions $\psi_\alpha$.

SUSY can be seen as a gauge symmetry that is required for a self-consistent theory of a particle with spin 3/2. Naively, elementary spin 1 particles $A_\mu$ contain negative-energy states, $A_0$ or $A_i$, which are, however, avoided by the existence of gauge symmetries. The self-consistency of elementary spin 2 particles $g_{\mu\nu}$ similarly requires gauging of the Poincaré symmetry, obtaining gravity. The self-consistency of spin 3/2 particles $\psi_{\mu\alpha}$ (where $\mu$ is a vector index and $\alpha$ a spinor index) similarly requires the existence of gauged supersymmetry, known as *supergravity*. Gravitinos are the spin 3/2 supersymmetric partners of the graviton.

The SM includes chiral fermions that are compatible with having an $N = 1$ supersymmetry, in which case chiral fermions fill *chiral super-multiplets* that pair spin 1/2 and spin 0 particles. Such supersymmetric theories can also contain gauge symmetries, resulting in *vector super-multiplets* that pair spin 1 vector gauge bosons with spin 1/2 *gauginos* $\lambda$. Chiral super-multiplets can have supersymmetric masses and interactions known as the *super-potential* terms.

A non-trivial feature of supersymmetric theories, crucial for the hierarchy problem, is that *the super-potential receives no perturbative quantum corrections.* Technically, loop quantum corrections from scalars running in the loop cancel against similar loops with fermionic super-partners. However, if supersymmetry were exact, the super-partners would have the same masses as the corresponding SM particles, which is excluded experimentally. To be phenomenologically viable, supersymmetry needs to be broken. The Higgs mass remains protected from unnaturally large quantum corrections, thus solving the hierarchy problem, if supersymmetry is broken 'softly' by the supersymmetric particles receiving additional electroweak-scale mass contributions, and thereby having weak-scale masses.

In the Minimal Supersymmetric Standard Model (MSSM) each SM particle is part of a super-multiplet and thus has a super-partner (these are known as *sparticles*). As already mentioned above, if the sparticles have weak scale masses, the hierarchy problem is solved. Furthermore, the RGE running of the SM gauge couplings gets modified and now allows for the SU(5) unification of the gauge couplings, at the scale $\approx 2 \; 10^{16} \, \text{GeV}$. This is below the Planck scale and can be above the bounds imposed by the absence of proton decay, depending on the details of the model (new supersymmetric dimension-5 contributions can be problematic in general).

SUSY pairs the Higgs doublet $H$ with a fermionic doublet partner, the Higgsino $\tilde{H}$, both of which then form a super-multiplet $\hat{H}$. Similarly, the lepton doublet $L$ is paired with a slepton $\tilde{L}$, forming a super-multiplet $\hat{L}$. The $\hat{H}$ and $\hat{L}$ supermultiplets have the same gauge quantum numbers, which means that one can write Yukawa-like interaction terms such as $\hat{L}\hat{Q}\hat{D}$, which would lead to unsuppressed lepton-flavour violating transitions, contrary to observations (in the SM these terms are not possible since $H$ is a scalar, while $L$ is a fermion). To avoid this and similar problems, theorists impose by hand a $\mathbb{Z}_2$ symmetry under which the supermultiplets housing the SM fermions are odd; thus $\hat{L} \to -\hat{L}$, but $\hat{H} \to +\hat{H}$. This $\mathbb{Z}_2$ symmetry can be combined with the $\mathbb{Z}_2$ under which all fermions are odd (including gauginos and higgsinos), $\psi \to -\psi$, to form the easier to remember *R-parity* under which all the SM fields are even and super-partners odd. Because of *R*-parity *the lightest super-partner (LSP) is stable and can be a DM candidate*. Among these, the neutralino was for a while considered to be so plausible that 'neutralino' was often used as synonymous with 'Dark Matter'.

Because of a rather rigid structure of the superpotential, SUSY requires two Higgs doublets in the MSSM. In general, this would result in problematic charge-breaking minima and flavour violations, while in SUSY this is avoided, because it predicts $H_u$ that couples to up quarks only, and $H_d$ that couples only to down quarks and to leptons, with a specific potential that avoids charge-breaking vacua. At tree level, the MSSM predicts that the SM-like Higgs is lighter than the $Z$ boson. A mildly heavier Higgs mass arises at loop level, especially, if sparticles are much heavier than the $Z$ boson.

The above appealing framework received a cold shower from the LHC: supersymmetry predicted a rich collider phenomenology, see section 7.2.1, while only the SM Higgs boson with mass $M_h = 125\pm0.17\,\mathrm{GeV}$ was found. In the MSSM such a Higgs mass, significantly above the mass of the $Z$ boson, is still possible, but requires stops (supersymmetric partners of the top quark) to be heavier than a few TeV. Together with the LHC exclusion bounds on sparticles, this means that SUSY can no longer fully keep the Higgs mass naturally smaller than the other heavier new-physics scales, significantly reducing the motivation for weak scale SUSY. In fact, this is a problem for most of the solutions to the hierarchy problem that postulate weak scale particles (large extra dimensions, composite Higgs), many of which have their own DM candidates. In the rest of this section we review different supersymmetric particles that could potentially be DM, and discuss what the implications of the (so far) negative LHC results are in each case. The LHC will keep running to accumulate more data at the same energy $\sqrt{s}$, which will increase the sensitivity to weakly coupled states such as the DM candidates discussed below, while in other cases (such as the production of new heavy colored particle) its discovery potential has to a large extend already been exploited.

## Neutralino: bino DM

The neutral *bino* $\tilde{B}$ is the fermionic SUSY partner of the hypercharge vector boson $B_\mu$. It mixes with the neutral component of *winos* (super-partners of $W_\mu^a$) and with *higgsinos* (super-partners of $H_{u,d}$), forming four mass eigenstates — the *neutralinos*.

The possibility that the lightest neutralino is mostly bino was often considered to be the most plausible SUSY DM candidate: *i)* various SUSY models predict this mass ordering, and *ii)* the observed DM cosmological abundance could be reproduced for a mass that matches that of natural weak scale SUSY, $M \sim M_Z$ [102]. Indeed, supersymmetric gauge interactions predict a Yukawa coupling between a bino $\tilde{B}$, a lepton $\ell$ and a slepton $\tilde{\ell}$ that is roughly equal to the hypercharge gauge coupling $g_Y$. A tree-level exchange of a slepton with mass $M_{\tilde{\ell}} \sim M$ induces a *p*-wave bino DM annihilation, with a cross section

$$\sigma v_{\mathrm{rel}}(\tilde{B}\tilde{B} \to \ell^+\ell^-) \sim v_{\mathrm{rel}}^2 \frac{g_Y^4}{4\pi M^2}. \tag{10.1}$$

The bino relic abundance matches the observed cosmological DM abundance for $\sigma v_{\mathrm{rel}}$ in eq. (4.13). This corresponds to $M \sim 120\,\mathrm{GeV}$, up to order one factors that depend on the number of light sleptons (1, 2

or 3) and on their 'chiralities' ($\tilde{\ell}_L$ or $\tilde{\ell}_R$).

After the LHC results this simple picture faces several obstacles. First of all, in SUSY models sleptons and binos are typically not much lighter than squarks and gluinos. The latter must now be heavier than $(1-2)\,\text{TeV}$, given that they are charged under QCD and would otherwise be copiously produced in the $pp$ collisions. One can still consider a heavier bino but then a larger prefactor in eq. (10.1) is needed in order to keep the bino annihilation cross section at its desired value. This can be achieved, thanks to the vast parameter space of SUSY models, but only in very specific corners of the parameter space:

○ The cross section is resonantly enhanced for $M$ close to $M_H/2$, where $M_H$ is the mass of one of the heavy Higgses.

○ The cross section gets enhanced, if some of the Yukawa couplings are larger than in the Standard Model. This is possible in supersymmetry, since there are two Higgs doublets, $H_u$ and $H_d$, and the ratio of two vevs, $\tan\beta \equiv \langle H_u\rangle/\langle H_d\rangle$, can be large.

○ Bino DM annihilation can get enhanced by co-annihilations with other supersymmetric particles, for example stops or sleptons, if these are not much heavier than the neutralino, $\Delta M \lesssim 100\,\text{GeV}$.

○ As we discuss in more details below, winos or higgsinos can be a thermal relic DM, if they are heavier than a TeV. Similarly, neutralino composed of just the right amounts of bino and wino and/or higgsino can be a thermal relic DM, for any sub-TeV mass [830].

## Neutralino: pure wino DM

*Winos* $\tilde{W}^a$ are the fermionic supersymmetric partners of the $\text{SU}(2)_L$ gauge bosons $W^a$, with $a = \{1, 2, 3\}$ and hypercharge $Y = 0$ [805]. The wino multiplet has three component, two charged winos $\tilde{W}^\pm$ (fermionic supersymmetric partners of the $W^\pm$ gauge bosons) and a neutral wino $\tilde{W}_3$ (partner of the $\text{SU}(2)_L$ gauge boson $W_3$, a linear combination of the photon and the $Z$). After supersymmetry-breaking, all three winos, $\tilde{W}_3$ and $\tilde{W}^\pm$, receive a common mass term $M_2$. After $\text{SU}(2)_L$-breaking, $\tilde{W}_3$ and $\tilde{W}^\pm$ mix with the other neutral and charged fermions, respectively. This mass mixing is negligible in the limit $M_2 \gg v$, where the winos $\tilde{W}_3, \tilde{W}^\pm$ form a quasi-degenerate $\text{SU}(2)_L$ triplet. Only the $\text{SU}(2)_L$ gauge interactions are then relevant for the wino DM phenomenology, such that in this limit $\tilde{W}_3, \tilde{W}^\pm$ constitute an example of a Minimal DM electroweak triplet with a zero hypercharge, see the discussion in section 9.3.4. In particular, wino DM has the desired thermal relic density for $M_2 \approx 2.5\,\text{TeV}$ (see table 9.2). This is much larger than $v$ and beyond the reach of the LHC (section 9.3.4). The predicted direct detection scattering cross section is below the present bounds. The most stringent constraints come from indirect detection, which excludes wino DM for cuspy DM profiles of the Milky Way, while it remains valid for non-cuspy ones.

## Neutralino: pure higgsino DM

*Higgsinos* are the fermionic supersymmetric partners of the two Higgs doublets $H_u$ and $H_d$. Decomposing the multiplets in their components, one gets charged $\tilde{H}^\pm$ and a complex neutral Higgsino $\tilde{H}_0$ (complex, since the multiplets carry a non-zero hypercharge $Y = \pm 1/2$). Higgsinos can have an $\text{SU}(2)_L$-invariant Dirac mass term $\mu$. If $\mu$ is much larger than the $\text{SU}(2)_L$-breaking mass terms, $\mu \gg v$, the higgsinos form a quasi-degenerate weak multiplet of mass $\mu$. In this limit, only the $\text{SU}(2)_L$ gauge interactions are relevant for higgsino DM phenomenology, such that higgsinos behave as a Minimal DM electroweak doublet with hypercharge $|Y| = 1/2$. Higgsino DM [831] has the desired thermal relic density when $\mu \approx 1.1\,\text{TeV}$ (see table 9.2). This is larger than $v$ and beyond the reach of LHC (section 9.3.4).

A pure Higgsino DM [831] is excluded since, due to $Y \neq 0$, it predicts direct detection scattering via tree level $Z$ boson exchanges, with a cross section orders of magnitude above the present bounds,

see fig. 5.3a. This can be rectified by the smaller $SU(2)_L$-breaking mass terms, leading to mass mixings with $\tilde{B}$ and $\tilde{W}_3$, and splitting the Dirac fermion $\tilde{H}^0$ into two non-degenerate Majorana fermions with modified couplings. As a result, a neutralino with a significant higgsino fraction remains excluded by direct detection bounds, but only up to fine-tuned 'blind spots', where different contributions to direct detection scattering interfere negatively, allowing for tuned cancellations [830].

## Sneutrino DM

As discussed above, $\hat{L}$ and $\hat{H}_u$ have the same gauge quantum numbers. The lepton doublets $\hat{L}$ contain *sneutrinos* $\tilde{\nu}_L$. These are a possible DM candidate [832], with the same gauge quantum numbers as the higgsinos, but with a different spin; they are spin 0 scalars instead of spin 1/2 fermions. Because of this difference, the thermal relic abundance of DM is reproduced for $m_{\tilde{\nu}_L} \approx 0.5\,\mathrm{TeV}$, in the $SU(2)_L$-symmetric limit (see table 9.2). As for higgsinos, the $Z$-mediated direct detection leads to strong exclusion bounds also for sneutrino DM. However, unlike the case of higgsino DM, the MSSM does not provide any immediate way to avoid direct detection bounds; one needs to add, e.g., right-handed sneutrinos with a Majorana mass of the appropriate size. Because of this, sneutrino DM attracted somewhat less interest recently.

## Gravitino DM

The *gravitino* is a hypothetical neutral spin-3/2 particle; it is the supersymmetric partner of the graviton, predicted by local supersymmetry. The gravitino has gravitational interactions analogous to the graviton, i.e. couplings suppressed by inverse powers of $M_{\mathrm{Pl}}$, so that it can be a gravitational DM candidate [100]. Its small interactions make it difficult to test it directly.

When supersymmetry is spontaneously broken at a scale $f$ (in a hidden sector, above the weak scale, for phenomenological reasons), an analog of the Higgs mechanism takes place: the gravitino $\Psi_\mu$ obtains a mass $m_{3/2} \sim f^2/M_{\mathrm{Pl}}$ by 'eating' the *goldstino* $\chi$, the 'Goldstone boson' of spontaneously broken supersymmetry. Hence the gravitino mass is governed by the SUSY-breaking scale, and can span a wide range of values. If gravity-mediation dominates all sparticle masses, all such masses are comparable. Naturalness suggests weak scale masses, obtained for $f \approx 10^{11}\,\mathrm{GeV}$, and no specific mass ordering is predicted: the gravitino could be the heaviest sparticle or the lightest one. Lower values of $f$ are allowed if sparticle masses receive weak-scale contributions from different dominant mechanisms (e.g. from gauge mediation): in such a case the gravitino is the lightest sparticle, as its mass only receives the gravity-mediated effect. In such a case, the gravitino mass could be $m_{3/2} \sim \mathrm{keV}$ if $f \sim 10^6\,\mathrm{GeV}$, or $m_{3/2} \sim \mathrm{MeV}$ for $f \sim 10^8\,\mathrm{GeV}$. Gravitinos much heavier than the weak scale can arise if super-symmetry does not solve the naturalness problem. In the 'unitary' gauge the action is written in terms of just the massive gravitino,

$$\mathcal{L}_\Psi = -\frac{1}{2}\varepsilon^{\mu\nu\rho\sigma}\bar{\Psi}_\mu\gamma_5\gamma_\nu\partial_\rho\Psi_\sigma - \frac{m_{3/2}}{4}\bar{\Psi}_\mu[\gamma^\mu,\gamma^\nu]\Psi_\nu + \frac{\bar{\Psi}_\mu S_{\mathrm{vis}}^\mu}{2\bar{M}_{\mathrm{Pl}}}, \tag{10.2}$$

where $S_{\mathrm{vis}}^\mu$ is the super-current composed of the visible sector fields. The longitudinal components of the massive gravitino inherit the interactions of the goldstino, and are thus suppressed by $1/f$ rather than by $1/M_{\mathrm{Pl}}$. Using the equivalence theorem, the gravitino production at high energies can be approximated as the sum of two channels: the first is the production of a massless gravitino and the second the production of the goldstino $\chi$, where for the latter the couplings to the visible sector are given by $\bar{\chi}(\partial_\mu S_{\mathrm{vis}}^\mu)/\sqrt{2}f$.

If supersymmetry-breaking is transmitted to the other sparticles by interactions that are stronger than gravity — for example through gauge interactions — the other sparticles will obtain larger SUSY-breaking masses than the gravitino, and thus the gravitino will be the lightest stable sparticle. To be a viable DM candidate, the gravitino also needs to match the observed cosmological DM abundance. This can happen in multiple ways, with the main contributions usually being:

- ○ *Thermal freeze out and decay* (see section 4.2.1). When the universe cools below the mass of the next-to-lightest supersymmetry particle (NLSP), $m_{\mathrm{NLSP}} > m_{3/2}$, the NLSP freezes out at its thermal relic abundance $\Omega_{\mathrm{NLSP}}$. At a later time, the NLSP decays to a gravitino and SM particles, so that the gravitino DM abundance is given by $\Omega_{3/2} = \Omega_{\mathrm{NLSP}} m_{3/2}/m_{\mathrm{NLSP}}$ (for simplicity we neglect entropy production, which is a good approximation, unless the NLSP decays very slowly, while dominating the energy density of the Universe) [833].

- ○ *Freeze-in* (see section 4.2.2). Gravitinos can be produced thermally, from scatterings of the particles in the plasma [152], such as gluon + gluon → gravitino + gluino. The space-time density of such scatterings at temperature $T$ is given by (at leading order in the strong coupling $g_{\mathrm{s}}$),

$$\gamma = \frac{T^6}{2\pi^3 \bar{M}_{\mathrm{Pl}}^2} \left(1 + \frac{M_3^2}{3m_{3/2}^2}\right) \frac{320.}{\pi^2} g_{\mathrm{s}}^2 \ln \frac{1.2}{g_{\mathrm{s}}}, \tag{10.3}$$

where $M_3$ is the gluino mass and $g_{\mathrm{s}}$ is the SU(3) gauge coupling. The second term in the parenthesis accounts for the enhanced coupling of the goldstino component of the gravitino. In this case, the gravitino production is dominated by the temperatures close to the reheat temperature $T_{\mathrm{RH}}$. Interestingly, the cosmological DM abundance can be reproduced for reasonable values of the parameters:

$$\Omega_{3/2}h^2 = 0.00167 \frac{m_{3/2}}{\mathrm{GeV}} \frac{T_{\mathrm{RH}}}{10^{10} \, \mathrm{GeV}} \frac{\gamma|_{T=T_{\mathrm{RH}}}}{T_{\mathrm{RH}}^6 / \bar{M}_{\mathrm{Pl}}^2}. \tag{10.4}$$

There are several ways that gravitino DM can be searched for. If SUSY were to be discovered at colliders and gravitino is the DM, this could be tested by searching for the (possibly slow) decays of the NLSP into the gravitino (detectable as missing energy) and a SM particle (for example, a photon, if the NLSP is the neutralino). Gravitino DM also does not have to be absolutely stable. For instance, it is slowly decaying in SUSY models with a slightly broken $R$-parity, resulting in indirect detection signals. Finally, the constraints on excessive cooling of stars and supernovæ sets bounds on ultra-light gravitinos [834].

## Axino and saxion DM

The axion is a speculative light boson that provides a solution to the strong CP problem, to be discussed in section 10.4. To form complete chiral supermultiplets requires two additional particles: the *axino* $\tilde{a}$, which is a fermionic superpartner of the axion, and the *saxion*, $\bar{a}$, which is its bosonic supersymmetric partner, required in order to match the number of bosonic and fermionic degrees of freedom. The QCD coupling of the axion super-multiplet is the supersymmetric extension of the axion coupling to gluons, see eq. (10.22). As in the non-supersymmetric case, the interactions are suppressed by the axion decay constant, $f_a$. In some SUSY models, the lightest supersymmetric particle can be either the axino or the saxion, which then can constitute in principle viable DM candidates [835]. The analyses of the cosmological abundance of the axino, as well as its observability, are similar to the case of the gravitino. For instance, in the same way as the gravitino, the axino can either be produced from the decays of the NLSP, giving,

$$\Omega_{\tilde{a}} = m_{\tilde{a}}\Omega_{\mathrm{NLSP}}/m_{\mathrm{NLSP}}, \tag{10.5}$$

or from thermal scatterings, giving

$$\Omega_{\tilde{a}}h^2 \approx 25 g_{\mathrm{s}}^6 \ln \frac{3}{g_{\mathrm{s}}} \frac{m_{\tilde{a}}}{\mathrm{GeV}} \frac{T_{\mathrm{RH}}}{10^4 \, \mathrm{GeV}} \left(\frac{10^{11} \, \mathrm{GeV}}{f_a}\right)^2. \tag{10.6}$$

Above, we chose a value of $f_a$ that is large enough such that the astrophysical bounds on axions are evaded (section 10.4). This then gives the correct relic abundance of GeV mass axino DM for relatively

low reheat temperature, $T_{\mathrm{RH}} \sim 10\,\mathrm{TeV}$ (and for appropriately higher $T_{\mathrm{RH}}$, if the axino is lighter).

### 10.1.3   Dark Matter and weak-scale extra dimensions

While weak scale supersymmetry has been considered by many as the most plausible solution to the Higgs mass hierarchy problem, a number of alternative scenarios have also been explored.

Compact extra dimensions can provide a solution to the hierarchy problem, since gravity spreads into a volume with more dimensions. The large four-dimensional Planck mass is therefore only an apparent scale, experienced by a four-dimensional observer, while the true quantum gravity scale, describing the interactions of higher dimensional gravity, can be much lower than the 4-dimensional Planck mass [836]. A tentative solution to the hierarchy problem would arise if the scale of quantum gravity could be pushed all the way down to the weak scale, while remaining in agreement with experimental data. Before the start of the LHC the constraints on these types of scenarios mostly came from indirect precision measurements, which may be evaded in special constructions. This rethinking of the hierarchy problem also lead to novel attempts at constructing viable DM candidates [105].

The most striking experimental consequence of extra dimensions is that any SM particle with access to extra dimensions would be accompanied by a tower of Kaluza-Klein (KK) excitations with mass splittings on the order of the compactification scale. The simplest model introduces one 5th dimension, a circle with radius $R$. If all SM particles freely propagate in it (a situation known as the *universal extra dimension* [836], their KK excitations have masses

$$M_n = n/R \tag{10.7}$$

with integer $n \geq 1$. Conservation of momentum in the 5th dimension becomes conservation of KK number $n$. *The lightest KK modes (LKP) with $n = 1$ would be stable, and could be a DM candidate.* Examples are the first KK mode of the photon (at tree level this tends to be the lightest, so it has been considered as the most plausible candidate and is arguably the one that has received most attention), or the first KK modes of the $Z$ boson, the Higgs, the neutrinos or even the gravitons[1].

This simplest model is not phenomenologically viable, since the would-be SM fermions — the $n = 0$ modes of the fermionic fields — are not chiral. The chiral SM fermion field content can be recovered if the extra dimension is an *orbifold*, i.e., by imposing a $\mathbb{Z}_2$ reflection symmetry on the extra dimension, transforming the circle into a segment with length $\pi R$. This breaks translation invariance, and thus also breaks the KK number, but leaves the $(-1)^n$ $\mathbb{Z}_2$ subgroup unbroken. This so called *KK parity* may or may not be broken by quantum corrections (see the discussion below). If KK parity is unbroken, the KK number can only change by an even number, and the lightest KK remains a stable DM candidate.

Taking the KK photon as an example, the total annihilation cross-section is approximately given by $\langle \sigma v \rangle \approx 0.3\,g^2/\pi M^2$ [105]. The observed cosmological abundance of DM is then reproduced for $1/R \approx 0.9\,\mathrm{TeV}$ [105], a value comparable to the bounds from electroweak precision data [105] available before the LHC. The collider phenomenology resembles loosely the phenomenology of the weak scale supersymmetry: colored KK excitations are pair produced with large QCD rates, resulting in events with jets and pairs of DM particles, which are then seen in detectors as missing energy signals. No signatures of the extra dimension were found at the LHC. The resulting bounds on the compactification scale, $1/R \gtrsim 2\,\mathrm{TeV}$ [105], exceed the value suggested by DM abundance, as well as the range motivated by a natural solution to the hierarchy problem.

Quantum corrections tend to generate kinetic and other interaction terms that are localized at the boundaries of the orbifold segment. Because the theory is non-renormalizable (the SM gauge, Yukawa, and scalar quartic interactions are no longer renormalizable in higher dimensions), such corrections are

---

[1]See however section 10.3.3 for a different realization of the latter case.

UV divergent. Generic localized terms would break the KK parity. In order to keep DM stable one needs to impose that also the terms on the borders of the orbifold segment respect the KK parity. Moreover, localized boundary terms in general also modify the KK masses, preventing one from predicting which KK excitation is the lightest and thus a possible DM candidate.

Non-minimal extra dimensional models allow significant flexibility: beyond the choices of the field content and the symmetries, there is now also the possibility of choosing the extra-dimensional 'geography'. On the flip side, this also adds a layer of arbitrariness not present in the four dimensional model building, rendering the constructions with stable lightest KK modes even more exceptional. Some KK parities are unavoidably broken by quantum anomalies, whose existence is a consequence of the chirality of the fermion zero modes [837]. Generically speaking, more extra dimensions imply more KK modes, and thereby stronger collider constraints on their sizes.

## 10.1.4 Dark Matter and composite Higgs

New strong interactions could be a solution to the hierarchy problem: in QCD, the low-energy scale $\Lambda_{\rm QCD} \sim 1$ GeV is naturally small thanks to the logarithmic running of the strong constant, and pions are parametrically lighter than this scale because they are pseudo-Goldstone bosons of a spontaneously broken approximate global symmetry, $\mathrm{SU}(2)_{q_L} \otimes \mathrm{SU}(2)_{q_R} \to \mathrm{SU}(2)_{q_V}$. Similarly, the weak scale could be naturally small, if the Higgs doublet is, in complete analogy with the QCD pions, a composite bound state of fermions (termed *techni-quarks*), bound together by a new strong interaction (termed *techni-color*, in analogy to color [838]) that confines around the weak scale. Technicolor models can result in a composite DM candidate, where in broad terms the discussion of composite DM models in section 9.5 still applies, but with the additional constraint that the model must also provide a composite Higgs at the weak scale. This additional requirement introduces several challenges:

1. As indicated above, the QCD pions are mildly lighter than the $\sim 1$ GeV QCD scale because they are the pseudo-Goldstone bosons of the accidental $\mathrm{SU}(2)_{q_L} \otimes \mathrm{SU}(2)_{q_R}$ chiral symmetry in the light quark sector, $q = \{u, d\}$, which is spontaneously broken by the $\langle \bar{q}q \rangle$ condensate to its vectorial subgroup $\mathrm{SU}(2)_{q_V}$. A similar $\mathcal{G} \to \mathcal{H}$ pattern of chiral symmetry breaking could have kept the Higgs mildly lighter than the new confining dynamics. However, the measured top and Higgs masses imply that the Higgs has sizeable interactions with the top quark and sizeable self-interactions, and thereby that the Higgs, apart from being light, does not exhibit the typical characteristics of a pseudo-Goldstone boson.

    ○ Within relatively complicated effective models the Higgs mass could still be kept naturally small, while allowing for a sizeable self-coupling of the Higgs and a large top quark Yukawa coupling, by introducing new symmetries. A well known example is *little Higgs with T-parity* where the new symmetry, *T-parity*, may also be responsible for DM stability [839]. In this case, the DM candidate is the *T*-odd partner of the hypercharge gauge boson or the *T*-odd partner of the neutrino.

2. The SM Higgs has a non-trivial pattern of Yukawa couplings to the SM quarks and leptons. It is difficult to imagine how such a structure can arise out of simple theories that only contain new fermionic techni-quarks with techni-color gauge interactions and masses. This problem can be bypassed, however, in ways that conflict with naturalness:

    ○ One possibility is to introduce an additional 4-fermion interaction among quarks and techni-quarks, which is suppressed by a new, unrelated scale. The stringent constraints on such interactions from flavor changing transitions can be satisfied, if the $\beta$-function for the renormalisation group evolution (RGE) of the technicolor gauge coupling is small (the tongue-in-cheek terminology is that the technicolor coupling is therefore *walking* [840], not running).

A warped extra dimension is conjectured to provide an approximate dual picture for such a type of strong dynamics [817].

- ○ The second possibility is to add elementary techni-colored scalars with their own Yukawa couplings [821]. Such models can accidentally conserve techni-baryon number, such that the lightest techni-baryon can be stable (similarly to the proton). If neutral, they can be a composite DM candidate, with specific couplings to the composite Higgs.

3. The interactions of composite particles are in general described by form factors. In the QFT language these correspond to a series of effective operators suppressed by the compositeness scale. Even if the new physics can be confined to the Higgs sector, dimension-6 operators such as $|H^\dagger D_\mu H|^2$ still affect the properties of the $W^\pm, Z$ bosons, which absorb 3 components of the Higgs doublet $H$ after the electroweak symmetry breaking. Already before the start of the LHC, the precision tests of $W^\pm, Z$ bosons were disfavouring natural technicolor theories, due to overwhelming agreement with the predictions of the SM, without any signs of new strong dynamics.

  - ○ Before the advent of the LHC a common approach was to shy away from explicit microscopic descriptions of compositeness, with the hope that the upcoming experimental data will shine new light on the problem. The focus was instead on effective theories that freely assumed special $\mathcal{G} \to \mathcal{H}$ breaking patterns of the accidental symmetry, which were devised such that they reduced conflicts with the precision data [841]. The assumed symmetry patterns could also lead to associated DM candidates [842].

4. Since natural composite Higgs models predict additional structure below the scale $4\pi v \sim 2\,\text{TeV}$, while none was observed at the LHC, the research in this direction effectively stopped to a screeching halt. Tuned models where the Higgs is composite at much higher scales remain viable, and could, for instance, have implications for axion DM [843].

## 10.1.5   Dark Matter and neutral naturalness

A possible reason for the non-observation of natural new physics at the LHC could be that none of the field content needed for Higgs naturalness is charged under the SM QCD. In such *neutral naturalness* models the production of new particles at the LHC would therefore be significantly suppressed. New colored particles lighter than about a TeV are typically excluded, while uncolored ones can still be allowed. Building models that satisfy such phenomenological requirements is not easy, because the largest coupling of the Higgs boson is to the top quark, which is colored and interacts with gluons (supersymmetry cancels the effects of the top quark on the Higgs mass by adding colored stops and gluinos).

An attempt in this direction are the *Twin Higgs* models in which the SM is accompanied by its mirror (possibly only partial) copy, such that the mirror $\mathrm{SU(3)}_c$ is not the ordinary QCD gauge group [844]. A discrete $\mathbb{Z}_2$ symmetry that exchanges the two sectors could protect the Higgs mass against leading radiative corrections, if it is a remnant of a larger spontaneously broken global symmetry, with the SM Higgs the pseudo-Goldstone boson whose potential, including the mass, is then tightly constrained due to the presence of the $\mathbb{Z}_2$ symmetry [844]. However, no such symmetry is present in the spectrum of observed particles, implying that the $\mathbb{Z}_2$ is broken. The protection of the Higgs mass against radiative corrections is thus only partial, and additional structure is required at energies above $\sim 10\,\text{TeV}$.

Much like the SM the (partial) mirror copy also possesses accidental symmetries: the mirror lepton number $L'$, and the mirror baryon number $B'$, as well as the conserved mirror charge $Q'$ (the mirror $\mathrm{U(1)}'_{\text{em}}$ may or may not be gauged, though). These conserved quantum numbers ensure the stability of the lightest set of particles in the mirror copy, the lightest lepton $\ell'$ and the lightest mirror baryon, which can then be the DM candidates. Depending on the details, the DM can be a weakly interacting

thermal relic (discussed in section 9.3.3) or asymmetric DM (sections 4.4 and 9.7). The latter possibility needs that the cosmological evolution resulted in an asymmetry in the population of DM particles in the mirror sector. If $U(1)'_{\text{em}}$ is gauged the Twin Higgs DM particles can also form dark atoms, which then has additional observational consequences (see section 9.5.3 for a general discussion).

### 10.1.6  Dark Matter and the relaxion

A number of different authors have proposed models in which the smallness of the squared Higgs mass is tied to the cosmological evolution. The general idea is that the point at which the squared Higgs mass in the Lagrangian becomes zero constitutes a critical threshold that separates qualitatively different physical situations. When the squared Higgs mass becomes negative, this results in a nonzero Higgs vacuum expectation value, $v$, which then induces nonzero masses for the SM fermions and the $W^{\pm}$, $Z$ gauge bosons. If one assumes that the squared Higgs mass is dynamically controlled by the vacuum expectation value of a new scalar singlet $\phi$, the cosmological evolution may dynamically select a small but non-vanishing value of $v$.

The first proposal along these lines is the so-called *relaxion* [845]. During inflation, the hypothetical relaxion scalar field $\phi$ is assumed to roll down its potential, while being coupled to the Higgs via a term $\sim g\phi|H|^2$. Because of this coupling the Higgs mass term evolves during the rolling phase: initially it is large and positive, while later it becomes small and negative, triggering the nonzero Higgs vacuum expectation value. Assuming that the relaxion has an axion-like coupling to gluons $\sim aG_{\mu\nu}\widetilde{G}^{\mu\nu}/f$ ($f$ is the equivalent of the axion decay constant, see section 10.4.2), then as soon as the SM quarks obtain the mass from the Higgs mechanism, the relaxion potential would receive a new periodic contribution $\sim \Lambda^3_{\text{QCD}} h\cos(\phi/f)$, typical of axion models (in phenomenologically viable version of the model a new group and new fermions are required not to be excluded experimentally). This oscillatory term generates, in the rolling slope of $\phi$, grooves whose depths increase as the rolling of $\phi$ proceeds, and eventually become valleys deep enough that the field $\phi$ gets stuck at one of the local minima. This halts the cosmological evolution of the relaxion field precisely at the point at which the squared mass term of the Higgs multiplet starts having a small negative value, thereby explaining the smallness of the weak scale.

While the mechanism works in principle, it also has some less appealing features. To make the idea viable the relaxion cannot be the QCD axion. Instead, a new confining sector and extra fermions are needed. The coupling $g$ needs to be extremely small, and thereby the relaxion field needs to be very light, spanning a large super-Planckian range of field values. The weak scale is naturally smaller than the cut-off of the relaxion theory, which, however, turns out to be much smaller than the Planck scale.

Various subsequent variants of the relaxion mechanism tried to improve the basic set-up. Here, we focus only on aspects relevant to DM. In the basic scenario, the relic relaxion particles cannot be the DM, since the relaxion mechanism occurs during the inflationary period, which then dilutes the relaxion abundance. Variants of the mechanism, in which relaxion could be the DM, have, however, also been explored [845]. One possibility, for instance, is that the relaxion DM is produced after reheating via the scatterings of SM particles in the plasma. This leads to DM in the keV range. Another possibility is that a high reheating temperature erases the trapping valleys, re-starting the rolling of the relaxion: when the temperature eventually decreases and the valleys reappear, another epoch of particle production occurs, creating the needed DM abundance (requiring that the relaxion is re-trapped in a close-by minimum guarantees that the Higgs mass does not significantly change after reheating). This leads to DM in the sub-eV range, with a phenomenology that resembles that of the axions (see section 10.4.5).

A different scenario attempting a cosmological anthropic solution to the hierarchy problem and predicting an ultra-light DM candidate is the so called *sliding naturalness* [845]. In these models anthropic selection results in a light Higgs, because a too large Higgs vacuum expectation value triggers an instability in a different light scalar, such that it rolls down its potential, and makes the cosmological constant big and negative, until it crunches the universe.

## 10.2   Dark Matter and cosmology

### 10.2.1   Dark Matter and Dark Energy: Chaplygin gas

At present, roughly 70% of the energy budget of the Universe is due to Dark Energy (DE), see eq. (C.11) in appendix C. The question why this energy density is so small, $V \sim \text{meV}^4 \sim 10^{-120} M_{\text{Pl}}^4$, is the so called *cosmological constant naturalness problem*. Numerically, it is even more severe than the Higgs mass hierarchy problem that we discussed in the previous section. Tentative natural solutions often involve new ultra-light scalars, which could be DM candidates, but also face serious theoretical obstacles as discussed in [846].

A concrete such attempt of unifying DM and DE is the so called *Chaplygin gas model of cosmology* [847]. At the level of the effective description, a Chaplygin gas is defined to be a fluid that has the following equation of state

$$\wp = -V^2/\rho. \tag{10.8}$$

The energy conservation equation $dU = -\wp \, d\mathcal{V}$, i.e., $d\rho/da = -3(\rho+\wp)/a$ is solved by $\rho = \sqrt{V^2 + B/a^6}$ where $B$ is an integration constant. The energy density of a homogeneous Chaplygin gas therefore initially scales as Dark Matter, $\rho \propto 1/a^3$, while at later times it behaves as a cosmological constant, $\rho \simeq V$. The observed current Dark Energy is reproduced by choosing a very small $V \sim 10^{-120} M_{\text{Pl}}^4$.

In the intermediate regime, $\rho$ differs from the $\Lambda$CDM model. This regime is tested by precise cosmological observations: the Chaplygin gas model of cosmology is excluded because its adiabatic sound speed $v_s^2 = \partial \wp / \partial \rho$ is not small enough to behave as DM during structure formation, as primordial inhomogeneities $\delta_k$ evolve as $\ddot{\delta}_k = -(k v_s/a)^2 \delta_k + \cdots$ (see section 1.3.1) [847]. CMB and LSS data demands $v_s^2 \lesssim 10^{-5}$, excluding $v_s^2 \sim 1$ [28].

The exclusion can be avoided by assuming a '*generalized Chaplygin gas*' with equation of state

$$\wp = -V^{1+\alpha}/\rho^{\alpha}, \tag{10.9}$$

such that $\rho = [V^{1+\alpha} + B/a^{3(1+\alpha)}]^{1/(1+\alpha)}$ still interpolates between DM and DE [847]. In the limit $\alpha \to 0$ this reduces to a combination of DM and DE, as in $\Lambda$CDM, so that the exclusion bounds are satisfied for small enough $\alpha \lesssim 0.2$. Of course, the Chaplygin gas model, as well as any unified DM/DE model which behaves at late times as pure DE, faces the difficulty of explaining the small scale (galactic and cluster) effects attributed to DM. For an attempt to address this difficulty, see e.g. Arbey (2006) in [847] and references therein.

At the fundamental level, a Chaplygin gas can arise from a '*branon*' — a scalar $\varphi$ motivated by string theory, which parametrises the position of a brane in an extra dimension — with kinetic action of the Born-Infeld type

$$\mathscr{L} = -V \sqrt{1 - \frac{(\partial_\mu \varphi)^2}{M^4}}. \tag{10.10}$$

Thanks to the higher-dimensional covariance, this action unifies the kinetic and potential energy into a Lorentz-like factor. In the homogeneous limit $\varphi(t)$ the corresponding Hamiltonian energy density is $\rho = V/\sqrt{1 - \dot{\varphi}^2/M^4}$, and the pressure is $\wp = \mathscr{L}$, giving the equation of state in eq. (10.8). The generalized Chaplygin gas could similarly be derived from a less motivated action

$$\mathscr{L} = -V \left[ 1 - \left( \frac{(\partial_\mu \varphi)^2}{M^4} \right)^{(1+\alpha)/2\alpha} \right]^{\alpha/(1+\alpha)} \simeq \frac{V}{M^4} \frac{(\partial_\mu \varphi)^2}{2} - V + \cdots . \tag{10.11}$$

For $\alpha \to 0$ this reduces to the case of having a sum of vacuum energy and DM contributions, as in the right-hand side of eq. (10.11), since then the additional terms become negligible. See [848] for related

attempts at getting DM and dark energy from rolling scalars.

More general interactions between DM and dark energy are discussed in section 8.4.3.

## 10.2.2 Dark Matter and inflation

Ignoring the cosmological constant problem for the moment, the ΛCDM model of cosmology still appears to require two extensions to the Standard Model: DM and inflation. Could the two have the same microscopic origin? In fact, microscopic theories that combine the two are rather easy to write down, but unfortunately also tend to not give any new easily observable consequences [849].

As a simple example let us take the scalar singlet DM model of section 9.2.1, where a scalar singlet DM candidate $S$ is added to the SM, but now let us require for $S$ to also act as the inflaton. This is possible, if $S$ has a non-minimal coupling to gravity, $\xi_S S^2 R/2$. Together with the quartic potential for $S$ in eq. (9.5), it is then possible to satisfy the conditions for slow roll inflation. At low energies, the theory remains invariant under the $S \to -S$ symmetry, even in the presence of a nonzero $\xi_S$ coupling, so that $S$ particles are stable and can act as DM. During inflation, however, this symmetry plays no role, and is broken by $\langle S \rangle \neq 0$. It gets restored once the period of inflaton ends and $S$ reaches the minimum at $S = 0$. After the end of inflation the system thermalises, resulting in a thermal bath that contains both the SM particles and DM particles. The stable $S$ particles later undergo thermal freeze-out, i.e., they behave exactly as in the singlet DM model of section 9.2.1.

Going beyond the minimal version of the scalar singlet model, one can change what the acceptable values for the parameters of the model are. For instance, if DM is produced via an alternative mechanism, such as the freeze-in (see section 4.2.2), then larger $S$ masses can also lead to acceptable DM abundance [849].

More involved models have also been considered, such as in Ballesteros et al. (2017, 2019) [849], which features, among other ingredients, a complex scalar field whose components play the role of the inflaton and of axion DM.

## 10.3 Dark Matter and string theory?

Einstein gravity is non-renormalizable and becomes strongly coupled at energies around the Planck scale. String theory is a possible extension of QFT that brings quantum gravity under control. However, a self-consistent description of strings as the quantum theory of gravity employs super-symmetry and 6 extra dimensions (or possibly 7: the string theory itself could be a limit of a deeper unknown theory). The extra dimensions should not be observable at energies we have probed so far; they need to be compactified in order to obtain the effectively $3 + 1$ dimensional space-time that we observe at energies well below the string scale. The resulting effective QFT, characterized by the choice of the gauge group, matter content, and values of the couplings, depends on the shape and geography of the compactification. Neither of these is known, either theoretically or experimentally, with an exponentially vast number of viable options, sometimes estimated as $10^P$ with $P$ comparable to the number of pages in this review.

Because of the abundance of choices, extracting any definitive predictions is exceedingly difficult, if not nearly impossible. The existence of a vast **landscape** of different vacua may, however, change our perspective on the two hierarchy problems discussed above. It allows to interpret the apparent unnaturalness of the vacuum energy and of the Higgs mass as being the results of **anthropic selection** within a multiverse, populated through cosmological inflation [850]. If taken seriously, this alternative then reduces the theoretical motivation for the natural solutions to the hierarchy problems discussed in the previous sections: maybe the value of the cosmological constant and the mass of the Higgs are simply unnaturally small and building theories of new natural physics at the weak scale is a misguided effort.

On the other hand, extracting testable implications for Dark Matter (or for any concrete structure

of the effective QFT) from theories with a large landscape of vacua, such as the string theory, is rather difficult.

An important issue is whether the existence of a DM abundance, that should be comparable to the observed one, is an anthropic necessity [851]. As discussed in section 1.3, in the cosmological evolution of our Universe DM facilitated the formation of large scale structures and the galaxies, since it started to cluster after the time of matter/radiation equality. At that time, normal matter could not cluster, since it was still coupled via the $e$ and $p$ charges to photons, which cannot cluster since they are relativistic. In our Universe, matter later fell into the inhomogeneities formed by the DM. Without DM, the ordinary baryons would have started clustering only after decoupling. Since the decoupling occurred not too long after the matter/radiation equality (see fig. C.1), the extra damping would have only moderately suppressed the matter power spectrum $P(k) = 2\pi^2 \delta_k^2 / k^3$. However, this change may have been enough to have had severe consequences. The bottom-right panel in fig. 1.5 shows that the matter inhomogeneity $\delta_k$ would have remained somewhat below 1, thus suppressing galaxy formation and thereby the existence of 'life'.

Whether this implies that a DM abundance somewhat larger than the ordinary matter density is an anthropic necessity, is debatable. A fly in the ointment is that varying other parameters allows one to recover a universe with abundant structures. For instance, a universe without any DM, but with larger primordial inhomogeneities, $\delta \approx 10^{-4}$, would have reached $\delta_k \sim 1$, just as in our Universe. One can also contemplate a universe with (much) more DM than measured: for a fixed value of $\delta$, having more DM relative to normal matter than in our Universe would have allowed for a bigger value of vacuum energy, $V \lesssim \delta^3 (MY_{\rm DM})^4$, and still be compatible with formation of structure. The anthropic constraints on universes with large DM densities are, in fact, quite loose. While a higher DM density would have precluded the cooling of galaxies, disk fragmentation and formation of stars, and hence 'life', this would occur only for values of DM density that are several orders of magnitude above the observed one [851].

Below we describe a few more concrete attempts at deriving DM implications from string theory: the possible existence of moduli and the axiverse, string-motivated supersymmetry, and the swampland program. Let us also mention here the *asymptotically safe gravity* framework, in which one insists on achieving a fixed point with gravity in the far UV. The coupling with gravity can place constraints on the couplings in the dark matter model, so that some phenomenologically viable models are ruled out and others have a reduced parameter space [852].

## 10.3.1   Dark Matter and string moduli

Supersymmetric string compactifications often feature *moduli*, chiral super-fields that either have no super-potential or have a super-potential that is negligible. Supersymmetry forbids perturbative quantum corrections to the super-potential, therefore moduli persist also at the quantum level. The vacuum expectation values of the moduli are not predicted by the string theory, while, importantly, they do control the values of some of the couplings in the effective QFT, that is valid at low energies. When supersymmetry is broken at low energies, the moduli tend to acquire masses $m$ that are comparable to the supersymmetry breaking scale. The moduli decay gravitationally, with the decay rate $\Gamma \sim m^3/M_{\rm Pl}^2$. Since the decay rate decreases with $m$, a modulus that were light enough could thus provide a (metastable) candidate for a gravitational DM, section 9.8.

This does not work in practice, however, since supersymmetry must be broken above the weak scale. Had supersymmetry been broken around the weak scale, $m \sim M_Z$, the moduli would have affected cosmology in a problematic way: they behave as matter, with $\rho \propto 1/a^3$, and would have eventually dominated the energy abundance (see section 4.3.4). Once they decayed they would have reheated the universe, but only up to a temperature $T_{\rm RH} \sim (\Gamma M_{\rm Pl})^{1/2} \lesssim$ MeV, for $m \lesssim 50$ TeV, which is not high enough to be compatible with the BBN bounds. The moduli problem is avoided, if supersymmetry is broken sufficiently above the weak scale (in which case it is not a solution to the Higgs naturalness

problem). In this scenario, the moduli decays could contribute to DM production [853]. A related possibility is that the SM vacuum could arise from a compactification that breaks supersymmetry already at the string scale. Computational difficulties prevent studying such compactifications, that likely do not feature light moduli.

## 10.3.2 Dark Matter and heavy-scale (or split) supersymmetry

Since string theory appears to require super-symmetry, it is interesting to entertain the possibility that super-symmetry exists, but does not solve the Higgs hierarchy problem, so that the scale of supersymmetry breaking need not be around the weak scale. Furthermore, this no longer needs to be a unique scale: the spectrum of super-symmetric partners might be *split* into two sets with very different masses, the supersymmetric scalars at mass scale $\tilde{m}$, and the supersymmetric fermions at mass scale $\tilde{M}$. Different arguments lead to different guesses for the possible values of $\tilde{m}$ and $\tilde{M}$:

- The SU(5) gauge unification favours $\tilde{M} \sim 10^6\,\text{GeV}$ [854].

- Within the MSSM, the measured Higgs mass can be reproduced for super-symmetry lighter than $\tilde{m} \lesssim 10^{10}\,\text{GeV}$, although $\tilde{m} \sim \tilde{M} \sim M_{\text{Pl}}$ is allowed within $3\sigma$ uncertainties [854].

- Symmetries allow $\tilde{M} \ll \tilde{m}$, and a one-loop splitting $\tilde{M} \sim \alpha\tilde{m}/4\pi$ arises in some models [854].

- Split supersymmetry allows DM as a weak doublet or triplet, with DM abundance reproduced thermally for $\tilde{M} \sim$ TeV (see section 9.3.4). Such a case is appealing, as the Higgs mass can be reproduced for $\tilde{m} \lesssim 10^8\,\text{GeV}$, and SU(5) gauge unification can be achieved [854]. However, other cases are also possible, such as gravitino DM.

- Finally, a simple possibility is that the super-string landscape is statistically dominated by vacua with super-symmetry broken already around the Planck scale. This would correspond to $\tilde{m} \sim M \sim M_{\text{Pl}}$.

## 10.3.3 Dark Matter and the string swampland

Attempts of obtaining observable implications from string theory recently shifted to the so called *'swampland'* program: the hope that compatibility with string theory restricts in interesting ways the QFTs that describe physics below the string scale. Theories outside the string-theory landscape are said to be in the swampland. While there is no proof of resulting restrictions on QFTs, various conjectures have been proposed:

1. Gravity must be weaker than gauge interactions: particles with gauge charge $q$ must be heavier than $m \gtrsim qM_{\text{Pl}}$, such that extremal black holes with mass $M = QM_{\text{Pl}}$ and charge $Q$ evaporate [855]. This would mean that a class of DM candidates, the Planck-scale remnants of primordial black holes (see section 3.1.1), are incompatible with string theory.

2. The Planck scale limits the range of scalar field values. Otherwise, the QFT description is invalidated by the appearance of a tower of extra states that are light, if there are super-Planckian scalar field value variations [855].

3. Achieving the small measured vacuum energy $V \sim \text{meV}^4 \sim 10^{-120}M_{\text{Pl}}^4$ implies a large variation of some field [855]. It is however not clear what forbids the alternative possibility that the small $V$ arises as an accidentally tuned cancellation between large contributions, without needing any large field range.

Trusting the last two conjectures and combining them, it was conjectured that the implied light tower can only be the Kaluza-Klein modes of one gravitational extra dimension with size set by the vacuum energy, $R \sim V^{-1/4} \sim 1/T_0 \sim \mu$m (with some generosity in numerical factors) [855]. This conjectured string theory prediction gives a theory of DM: graviton KK modes cosmologically produced by UV-dominated gravitational freeze-in. The space-time density of scatterings that produce one KK graviton with mass $M$ from the SM plasma is $\gamma \sim T^6 e^{-M/T}/M_{\rm Pl}^2$. Summing over the number of accessible KK modes $\sim T_{\rm RH}R$, and taking into account that their typical mass is $M \sim T_{\rm RH}$ gives the DM abundance

$$\Omega_{\rm DM} \sim \frac{M}{T_0} \left.\frac{\gamma}{Hs}\right|_{T=T_{\rm RH}} \sim \frac{T_{\rm RH}^3}{M_{\rm Pl}T_0^2}. \tag{10.12}$$

The amount of gravitationally produced graviton KK can match the DM abundance assuming a low reheating temperature $T_{\rm RH} \sim$ GeV. However, the decay rate of GeV-scale KK gravitons into SM particles $\Gamma({\rm KK} \to \gamma\gamma, e^-e^+) \sim M^3/M_{\rm Pl}^2$ exceeds the bounds in fig. 6.15 by about 12 orders of magnitude. To avoid being excluded one can assume mildly broken translational invariance in the extra dimension, such that the produced GeV-scale graviton KK decay dominantly to lighter graviton KK with masses $M \sim 10\,$keV, heavy enough to satisfy bounds on warm DM, and light enough to be satisfy DM stability bounds [855].

Furthermore, the black holes in 5 dimensions can be alternative DM candidates, even if they are lighter than $10^{-15}\,M_\odot$ (unlike in fig. 3.3b), since they emit less Hawking radiation [855].

### 10.3.4   Dark Matter and string axiverse

Compactifications of the extra dimensions in string theory tend to predict the existence of very light pseudo-scalar fields that, from the low energy perspective, behave as axion-like particles (see section 10.4). In string theory these are the $n = 0$ massless KK modes of antisymmetric tensor fields, with masses protected by the higher dimensional gauge symmetry to all orders in perturbation theory, while nonzero contributions to their masses do arise from non-perturbative phenomena (in a similar way as for the QCD axion, see section 10.4.2). However, unlike the QCD axion, which is usually assumed to be just a single light pseudo-Nambu-Goldstone boson, the generic expectation from the string theory is that there are many such light pseudo-scalar fields, possibly even populating each decade of mass all the way down to the Hubble scale of $m_a \sim 10^{-33}\,$eV, a hypothesis that is often referred to as the *string axiverse* [856]. Depending on the details of the string axiverse DM could then be composed from one or several different axions with differing masses (unlike the discussion in the next section, where for the most part we will assume that there is a single axion).

## 10.4   Axion Dark Matter

The axion is a hypothetical light pseudo-scalar postulated in 1977 in order to explain dynamically why the CP violation in strong interactions is very small (see section 10.4.1 below). Interestingly, the axion can be a viable ultra-light DM candidate (section 3.4), possibly produced via the misalignment mechanism (section 4.3.4). This section provides further details specific to axion DM. Importantly, axion DM can be probed via its effective interactions, which are discussed in section 10.4.2. The main classes of renormalizable axion models are summarized in section 10.4.3, while in section 10.4.4 we discuss the expected cosmological relic density of axions. In section 10.4.5 we summarize experimental efforts for detecting axion DM.

## 10.4.1 The QCD $\theta$ angle and the chiral anomaly

This topic involves aspects of advanced quantum field theory that cannot be presented in a simple way: chiral anomalies and extended field configurations with topological charges. We thereby just state the main results, see [857, 858] for additional extensive discussions.

The symmetries and field content of the SM allow for one extra CP-violating term in the SM Lagrangian, parameterized by a constant $\theta$,

$$\mathscr{L} = \mathscr{L}_{\text{SM}} - \theta \frac{g_{\text{s}}^2}{32\pi^2} G_{\mu\nu}^a \tilde{G}_{\mu\nu}^a, \tag{10.13}$$

where $g_{\text{s}}$ is the QCD gauge coupling, $G_{\mu\nu}^a$ is the gluon field strength and $\tilde{G}_{\mu\nu}^a \equiv \frac{1}{2}\epsilon_{\mu\nu\alpha\beta}G_{\alpha\beta}^a$ its dual. Naively, this extra term is irrelevant since it is a total derivative:[2]

$$G_{\mu\nu}^a \tilde{G}_{\mu\nu}^a = \partial_\mu K^\mu \qquad \text{with} \qquad K^\mu = \epsilon^{\mu\alpha\beta\gamma}C_{\alpha\beta\gamma}, \qquad C_{\alpha\beta\gamma} = A_\alpha^a G_{\beta\gamma}^a - g_{\text{s}}f^{abc}A_\alpha^a A_\beta^b A_\gamma^c/3. \tag{10.14}$$

Using Gauss's theorem such a total derivative term in the action can be rewritten as a boundary contribution at Euclidean infinity. Were $K^\mu$ to vanish sufficiently quickly at infinity, the term could be dropped. However, this is not the case for all gauge configurations in a non-abelian gauge group such as QCD. Instead, it is equal to the Cartan-Mauer topological invariant of the mapping $g(\hat{x}_\mu)$, which maps $\hat{x}_\mu$ parametrising the boundary (i.e., the $x^\mu$, but at Euclidean infinity, and therefore just a direction) into an element $g$ of the SU(3) gauge group. As a topological quantity it is quantized, and given by

$$\frac{g_{\text{s}}^2}{32\pi^2} \int d^4x \ G_{\mu\nu}^a \tilde{G}_{\mu\nu}^a = q, \tag{10.15}$$

where $q = \{0, \pm 1, \pm 2, \ldots\}$ counts how many times the hypersphere at infinity wraps around the group manifold (this wrapping has an orientation, and thus $q$ can be either negative or positive). An 'instanton' field configuration corresponds to $q = 1$, and is given by

$$G_\mu^a T^a = ig_{\text{s}} \frac{r^2}{r^2 + R^2} g^{-1}(x)\partial_\mu g(x), \qquad g(x) = \frac{x_4 + 2ix_i\sigma_i}{r}, \tag{10.16}$$

where $R$ is a constant controlling the size of the instanton, and $\sigma_i$ are the Pauli matrices of the SU(2) subgroup of SU(3). For $r \to \infty$ this field configuration is a pure gauge.

Such field configurations give non-perturbative corrections to the action, though exponentially suppressed by a factor $e^{-8|q|\pi^2/g_{\text{s}}^2}$. They are important in the case of QCD, since the gauge coupling $g_{\text{s}}$ becomes non-perturbatively large for energies around the QCD scale, $\Lambda_{\text{QCD}} \approx 250\,\text{MeV}$.

The QCD $\theta$ term is connected to the chiral anomaly. The SM Yukawa couplings $y$, which lead to quark masses after electroweak symmetry breaking are, in Weyl notation, given by,

$$\mathscr{L} \supset y_U \, UQH + y_D DQH^* + \text{h.c.} \supset m_u u_L u_R + m_d d_L d_R + \text{h.c.}. \tag{10.17}$$

Generational indices are left implicit, e.g. $y_U \, UQH$ denotes $\sum_{ij}(y_U)_{ij} U_i Q_j H$. The Yukawa couplings are

---

[2]In a more geometrical language, the $\theta$ term in eq. (10.13) can also be written as $F = \epsilon^{\mu\nu\rho\sigma}F_{\mu\nu\rho\sigma}$, where $F_{\mu\nu\rho\sigma} = \partial_{[\mu}C_{\nu\rho\sigma]}$ is the 4-form that follows from the Chern-Simons 3-form $C_{\nu\rho\sigma}$. This can be seen as a QCD composite, where the QCD gauge invariance also induces a 3-form gauge invariance on $C$. The discussion about the role of the QCD axion in solving the strong CP problem can then be rephrased, in a more abstract language, by noticing that a 3-form in 3+1 dimensions is analogous to a 1-form (a vector) in 1+1 dimensions; the equations of motion are solved by a constant $F$, which is the 1+1 dimensional analog of the electric field. In the same way as a Higgsed electric field in 1+1 dimension is screened, the axion screens the 3-form by putting it in its Higgs phase.

in general complex, but are usually made real and positive by performing axial phase redefinitions of the quark fields $q = \{u, d\}$:

$$q_L \to e^{i\delta_q} q_L, \qquad q_R \to e^{i\delta_q} q_R, \qquad \text{i.e., in Dirac notation} \qquad \Psi_q = \begin{pmatrix} q_L \\ \bar{q}_R \end{pmatrix} \to e^{-i\gamma_5 \delta_q} \Psi_q \qquad (10.18)$$

where $\gamma_5 = \mathrm{diag}\,(-1, -1, 1, 1)$. At the quantum level this symmetry is anomalous since the measure of the path integral is not invariant under chiral redefinitions of the phase of the quark fields:

$$\mathscr{D}\Psi_q \mathscr{D}\bar{\Psi}_q \to \mathscr{D}\Psi_q \mathscr{D}\bar{\Psi}_q \,\exp\left[ i \frac{\alpha_{\mathrm{s}}}{4\pi} \alpha_q \int d^4 x\, G^a_{\mu\nu} \tilde{G}^a_{\mu\nu} \right]. \qquad (10.19)$$

This means that the phase redefinition of quarks changes the $\theta$ term as

$$\theta \to \theta - 2(\delta_u + \delta_c + \delta_t + \delta_d + \delta_s + \delta_b). \qquad (10.20)$$

This means that $\theta$ itself is not physical, and instead the invariant phase combination $\bar{\theta} \equiv \theta + \arg \det y_U y_D$ is a physical parameter of the SM.

One physical consequence of the fact that the axial quark phase rotations are not a symmetry of the QCD action, is that the corresponding would-be Goldstone boson of spontaneous symmetry breaking of the approximate flavor group in the light quark sector — the $\eta'$ meson — acquires a QCD-scale mass of about a GeV, in agreement with data.[3] Another physical consequence is that, since $\bar{\theta}$ is a physical parameter that violates CP, it contributes to the *electric dipole moment of the neutron*, giving [859]

$$d_n \approx \bar{\theta} e m_\pi^2 / m_N^3 \approx 10^{-16}\, \bar{\theta}\, e\, \mathrm{cm}. \qquad (10.21)$$

The experimental bound $|d_n| \lesssim 1.8\ 10^{-26} e\, \mathrm{cm}$ [860] implies $|\bar{\theta}| \lesssim 10^{-11}$, which raises the question of why $\bar{\theta}$ is so small.[4] This could have had a very simple solution, had the up quark been massless. In that case the SM Lagrangian would have had a U(1) symmetry, which would have allowed to freely choose the phase $\delta_u$ such that $\bar{\theta} = 0$ (that is, in this case $\bar{\theta}$ would no longer have been a physical quantity and one could have simply set $\bar{\theta} = 0$). However, this is not the world we live in; all the quarks are massive and the CKM matrix contains a large CP-violating phase. Without further tunings one would therefore expect that the phases $\delta_q$ in the quark diagonalization are of order one, and so would $\bar{\theta}$ be, in stark contrast with observations. Axions are a possible solution to this conundrum.

## 10.4.2   The axion effective Lagrangian

The basic idea underpinning the axionic solution to the strong CP problem, allowing to explain the smallness of the $\bar{\theta}$ term, is the assumption that the full Lagrangian (i.e., the SM supplemented by additional fields) is invariant under a global U(1) symmetry, such that at least some of the quark mass phases $\delta_q$ can be freely adjusted. As in the massless up quark example above, the $\bar{\theta}$ parameter is then no longer observable and can be set to zero. However, since all the quarks are in fact massive, such a U(1) symmetry must be spontaneously broken at some high scale, leaving at low energies the SM and a light

---

[3]More precisely, the axial quark phase rotations that have no flavor diagonal component, such as $\delta_u = -\delta_d$, and $\delta_u = \delta_d = -\delta_s/2$, do not change $\theta$, see eq. (10.20). The low energy QCD Lagrangian for light quarks thus has an approximate $\mathrm{SU}(3)_L \otimes \mathrm{SU}(3)_R$ flavor symmetry (broken explicitly by small quark masses, but not by the QCD anomaly). This is spontaneously broken to vectorial $\mathrm{SU}(3)$, with $\pi$ and $K$ mesons the corresponding pseudo-Nambu-Goldstone bosons. The flavor diagonal axial rotation, $\delta_u = \delta_d = \delta_s$, however, is anomalous, so that the corresponding would be pNGB, $\eta'$, does not have its mass protected by a symmetry.

[4]The SM contains also contains the CP-violating CKM phase, that contributes to the neutron electric dipole moment as $d_n \approx 10^{-32} e\, \mathrm{cm}$, suppressed by higher loops and CKM angles.

Goldstone boson from the U(1) breaking, the axion $a$.

In the next subsection we will discuss the concrete axion models and identify the U(1) symmetries that get spontaneously broken. First, however, let us discuss the effective interactions of the axion, which can be derived from model-independent considerations. Having assumed that $a$ is a Goldstone boson, its Lagrangian is invariant under shifts $a(x) \to a(x) + \text{constant}$, up to anomalies. The axion therefore only has derivative couplings and anomalous couplings to gauge vectors. At energies above the weak scale the $\text{SU}(2)_L$-invariant axion Lagrangian is given by

$$
\begin{aligned}
\mathscr{L}_{\text{axion}} \;=\;& \mathscr{L}_{\text{SM}} + \frac{(\partial_\mu a)^2}{2} - \frac{a}{f_a}\left[\frac{\alpha_{\text{s}}}{8\pi}G^a_{\mu\nu}\tilde{G}^a_{\mu\nu} + c_2\frac{\alpha_2}{8\pi}W^a_{\mu\nu}\tilde{W}^a_{\mu\nu} + c_1\frac{\alpha_Y}{8\pi}B_{\mu\nu}\tilde{B}_{\mu\nu}\right] + \\
&+ \frac{\partial_\mu a}{f_a}\left[c_H(H^\dagger iD_\mu H) + \sum_i c_i(\bar{\psi}_i\gamma_\mu\psi_i) + \text{h.c.}\right].
\end{aligned}
\tag{10.22}
$$

Here, $H$ is the Higgs doublet; the Weyl spinors $\psi_i$ describe all the SM fermions; $f_a$ is known as the 'axion decay constant'; while $c_i$ are dimension-less constants that parameterize the axion derivative couplings. In general, the SM fermion mass terms might have axion-dependent phases: axion-dependent redefinitions of the quark fields, as in eq. (10.18), allow to eliminate such terms in favour of anomalous couplings and shifts in $c_H, c_i$, so that all axion couplings are contained in $\mathscr{L}_{\text{axion}}$ above. Using the fact that vector currents are conserved, $\partial_\mu\bar{\Psi}_i\gamma^\mu\Psi_i = 0$, the interactions with the SM fermions in (10.22) can be rewritten in terms of couplings to just the axial currents of the SM fermions, $\partial_\mu a(\bar{\Psi}_i\gamma_\mu\gamma_5\Psi_i)/f_a$ (with $\Psi_i$ now Dirac fermions). In eq. (10.22) we assumed flavor-diagonal couplings to the SM fermions (i.e. flavour-universal U(1) charges); flavor-violating couplings may be possible as well, and will be discussed in section 10.4.5.

The Higgs obtains a weak-scale vacuum expectation value: by making use of an $a$ dependent hypercharge transformation one can remove axion/Higgs mixing and couplings. After integrating out the higgs $h$ and the heavy gauge bosons $W, Z$, one obtains the UV contribution to axion coupling to photons,

$$
-\frac{a}{f_a}\frac{\alpha}{8\pi}c_{\gamma,\text{uv}}F_{\mu\nu}\tilde{F}_{\mu\nu}, \qquad c_{\gamma,\text{uv}} = c_1 + c_2.
\tag{10.23}
$$

At low energies the axion-gluon couplings will generate an additional contribution to axion coupling to photons, see eq.s (10.37), (10.24) below. Furthermore, to obtain the effective interactions of axion with fermions at low energies, heavy fermions can be integrated out by simply dropping the corresponding currents in (10.22), since we work in a basis where fermion masses do not depend on the axion.

At the QCD scale, strong interactions become non-perturbative and generate an axion potential. Since strong interactions respect CP, the axion potential is even in $\bar{\theta} + a/f_a$, and thereby has a CP-invariant extremum, which Vafa and Witten proved to be a minimum [861]. The axion thus acquires dynamically a vacuum expectation value that exactly cancels the CP-violating $\theta$ term (up to very small CP violating contributions from the SM weak interactions), explaining the smallness of $\bar{\theta}$.

To compute the axion mass generated by the coupling to the QCD anomaly, it is convenient to trade the $aG\tilde{G}$ term for couplings of $a$ to quarks. This can be done by performing a phase redefinition of the light quark fields,

$$
\Psi_q \to e^{-ia(Q_V + Q_A\gamma_5)/f_a}\Psi_q, \qquad \text{with} \qquad \text{Tr}\,Q_A = \frac{1}{2},
\tag{10.24}
$$

where $\Psi_q = \{\Psi_u, \Psi_d, \Psi_s\}$, is a vector of light quark Dirac spinors. This shifts the $a\gamma\gamma$ coupling to $c_\gamma = c_{\gamma,\text{UV}} - 4\text{Tr}\,(Q_A Q^2_{\text{em}})$ (the trace gives a factor of 3 when acting on quarks) and changes the light quark mass matrix $M_q = \text{diag}(m_u, m_d, m_s)$ to $e^{iaQ_A/f_a}M_q e^{iaQ_A/f_a}$. In this picture, the quark condensate $\langle\bar{q}q\rangle \neq 0$ generates an axion potential. Using the standard chiral perturbation theory tools, the phase-shifted $\bar{\Psi}_q M_q \Psi_q$ mass term is converted into an effective Lagrangian for the lightest mesons: pions, kaons and the $\eta$ meson. By choosing $Q_A = M_q^{-1}/2\text{Tr}\,[M_q^{-1}]$ one avoids the pion/axion mixing and then rather straightforwardly obtains the expression for the axion mass, which is at leading order in chiral expansion

is given by,

$$m_a = \frac{m_\pi f_\pi / f_a}{\sqrt{(1 + m_u/m_d)(1 + m_d/m_u + m_d/m_s)}} \approx 0.57 \, \mathrm{meV} \frac{10^{10} \, \mathrm{GeV}}{f_a}, \tag{10.25}$$

where $f_\pi = 93 \, \mathrm{MeV}$ is the pion decay constant. One can similarly obtain the axion couplings to pions and nucleons, which are also suppressed by the axion decay constant, $f_a$. This is a large supression; bounds from axion search experiments and star cooling imply $f_a \gtrsim 10^9 \, \mathrm{GeV}$ (see section 10.4.5).

### 10.4.3   Axion models and theories

In 1977 Peccei and Quinn (PQ) proposed the original axion model [857], in which they extended the Higgs sector of the SM by introducing two Higgs doublets instead of a single one in the SM; one coupled to the up quarks and one coupled to the down quarks,

$$\mathcal{L} \supset y_U U Q H_u + y_D D Q H_d + \mathrm{h.c.} + V(H_u, H_d). \tag{10.26}$$

Postulating that no $H_u H_d$ nor $(H_u H_d)^2$ terms are present in the Higgs potential, the Lagrangian has a $\mathrm{U}(1)_{\mathrm{PQ}}$ global symmetry: $H_{u,d} \to e^{-2i\alpha} H_{u,d}$ and $q \to e^{i\alpha} q$ for $q = \{Q, U, D\}$, implying $\mathrm{U}(1)\,\mathrm{SU}(3)_c^2$ anomalies. The $\mathrm{U}(1)_{\mathrm{PQ}}$ is spontaneously broken by the Higgs vacuum expectation values, resulting in a light pseudo-Nambu-Goldstone boson — the axion. In this case the axion has a weak-scale decay constant $f_a \approx v = 174 \, \mathrm{GeV}$, a possibility that is experimentally excluded. Subsequently, more complicated axion models with new heavier particles have been proposed, in which $f_a$ can be significantly larger. Since the couplings of the axion to visible matter, proportional to $1/f_a$, is now highly suppressed, these models are often also referred to as the *invisible axion* models. Broadly speaking they are of two types.

In their simplest version, the **Dine-Fischler-Srednicki-Zhitnitskii (DFSZ) axion models** [857] add to the PQ axion model an extra complex heavy scalar $\sigma$, assumed to be neutral under the SM gauge group. The theory is again assumed to be symmetric under a global $\mathrm{U}(1)_{\mathrm{PQ}}$ such that the $H_u H_d$ and $(H_u H_d)^2$ terms are absent, but now also $\sigma$ transforms under the $\mathrm{U}(1)_{\mathrm{PQ}}$ symmetry:

$$\sigma \to e^{2i\alpha} \sigma, \qquad q \to e^{i\alpha} q \qquad H_u \to e^{-2i\alpha} H_u, \qquad H_d \to e^{-2i\alpha} H_d. \tag{10.27}$$

The term $\sigma^2(H_u H_d)$ is therefore allowed and is present in the Lagrangian. The $\mathrm{U}(1)_{\mathrm{PQ}}$ symmetry is broken by the vacuum expectation values of $H_u, H_d$, and $\sigma$. Assuming that $\sigma$ acquires a vacuum expectation value $f_a \gg v$, the pseudo-scalar in $H_{u,d}$ becomes heavy, and the light axion dominantly becomes the mode $a$ that parameterizes the phase of $\sigma$, i.e., $\sigma \approx f_a e^{ia/f_a}/\sqrt{2}$, and thus around the origin $a = \sqrt{2} \mathrm{Im}\,\sigma$. The main phenomenological gain over the original PQ model is that $f_a$ is a free parameter that can be large; the present bounds from axion searches are evaded, if the decay constant $f_a \approx \langle \sigma \rangle \gg v$ is large enough.

The simplest version of the **Kim-Shifman-Vainstein-Zakharov (KSVZ) axion models** [857] adds to the SM a vector-like heavy quark $\Psi_Q = (Q_L, \bar{Q}_R)$ and a heavy scalar $\sigma$, with all three for simplicity assumed to be neutral under electro-weak interactions, but charged under QCD. In order to forbid the Dirac mass term, $M_Q\, Q_L Q_R$ one can, for example, impose a symmetry

$$Q_L \to -Q_L, \qquad Q_R \to Q_R, \qquad \sigma \to -\sigma. \tag{10.28}$$

The mass of the heavy quark is then only due to the vacuum expectation value of $\sigma$,

$$\mathcal{L} = \mathcal{L}_{\mathrm{SM}} + \bar{\Psi}_Q i \slashed{\partial} \Psi_Q + |\partial_\mu \sigma|^2 + y\,\sigma Q_L Q_R + V(\sigma). \tag{10.29}$$

The U(1)$_{\text{PQ}}$ global symmetry

$$\Psi_Q \to e^{i\gamma_5 \alpha_Q} \Psi_Q, \qquad \sigma \to e^{-2i\alpha_Q} \sigma \tag{10.30}$$

is spontaneously broken by the vacuum expectation value of the $\sigma$, giving rise to a light axion $a$ with a large decay constant $f_a \approx \langle \sigma \rangle$. The $aG\tilde{G}$ coupling is the result of $\Psi_Q$ carrying a U(1)$_{\text{PQ}}$ charge, while the U(1)$_{\text{PQ}}$ is anomalous under QCD.

These minimal models assume an ad-hoc global PQ symmetry, without trying to understand its origin. Arguments based on properties of black holes suggest that theories which contain gravity cannot have exact global symmetries. In contrast, approximate global symmetries can easily arise as accidental symmetries in renormalizable theories. Two examples of such an accidental global symmetry are the baryon and lepton number in the SM, which simply follow from the assumed field content and gauge symmetries of the SM, without ever being explicitly imposed. Such accidental symmetries are expected to be broken by non-renormalizable effective operators of dimension $d > 4$, and thus suppressed by $1/\Lambda_{\text{UV}}^{d-4}$, where $\Lambda_{\text{UV}}$ is a large energy scale possibly comparable to the Planck mass, $\Lambda_{\text{UV}} \lesssim M_{\text{Pl}}$. Such contributions can lead to nonzero $\bar{\theta}$, yet are not observationally problematic, if they are small enough, $|\bar{\theta}| \lesssim 10^{-11}$ (in the same way as the SM electroweak corrections are not). Naively, this is the case, if the new contributions to the axion mass term in the axion potential, roughly of the size $\delta m_a^2 \sim f_a^2 (f_a/\Lambda_{\text{UV}})^{d-4}$, are about $10^{11}$ times smaller than the QCD contribution, $m_a^2 \sim \Lambda_{\text{QCD}}^4/f_a^2$. This implies a significant constraint: even assuming extremal values, $\Lambda_{\text{UV}} \sim M_{\text{Pl}}$ and $f_a \sim 10^9\,\text{GeV}$, the above estimates still imply that all the higher dimensional operators up to dimension $d \approx 9$ need to respect the PQ symmetry. An accidental symmetry that remains unbroken up to such high dimension is not a typical feature of quantum field theories. Quite the opposite — the more complicated operators of higher dimension typically break accidental symmetries. For example, the accidental symmetries of the renormalizable SM are broken at dimension 5 (lepton number and flavour) and 6 (baryon number). This so called **PQ quality problem** drastically reduces the set of plausible theories for a QCD axion. One may even wonder whether the whole setup is not flawed.

There are two possible resolutions to the PQ quality problem. On the one hand, we do not know for certain that gravity breaks global symmetries in a way that can be approximated by a series of Planck-suppressed operators. Computations based on Euclidean wormholes indicate that the violations of global symmetries are not power suppressed, and are rather much smaller, suppressed by $\exp[-\mathcal{O}(M_{\text{Pl}}/\sigma)]$. For $\langle \sigma \rangle = f_a \ll M_{\text{Pl}}$ this would then result in negligible corrections to the typical axion potential (though violations would be large in models where the axion radial mode $\sigma$ acquires Planckian values inside wormhole throats). On the other hand, even if breakings of the PQ symmetry really are just power suppressed, it is still possible to build viable theories of QCD axion, by introducing additional gauge symmetries that forbid U(1)$_{\text{PQ}}$ breaking operators to very high $d$.

## 10.4.4 The cosmological axion density

Different mechanisms can contribute to the cosmological relic density of axions. Cold axions with energy below the axion mass behave as DM [101], while hot axions behave as extra radiation.

First, axions belong to the class of DM candidates discussed in section 3.4: ultra-light scalar fields whose coherent oscillations contribute to the energy density of the universe. Their cosmological abundance, set by the *initial misalignment* mechanism discussed in section 4.3.4, is given in eq. (4.70). For the specific case of the axion, with the mass given in eq. (10.25), it reduces to

$$\Omega_a^{\text{mis}} \approx 0.15 \left( \frac{f_a}{10^{12}\,\text{GeV}} \right)^{7/6} \left( \frac{a_*}{f_a} \right)^2. \tag{10.31}$$

The power 7/6 (rather than 1) can be derived from the formalism of section 4.3.4, taking into account that at $T \gtrsim \Lambda_{\text{QCD}}$ the axion mass gets suppressed as $m_a(T) \sim m_a(\Lambda_{\text{QCD}}/T)^4$ (the temperature $T_*$ at

which $m_a \sim H$ is just above $\Lambda_{\mathrm{QCD}}$ for $f_a \sim 10^{12}\,\mathrm{GeV}$).

Assuming that axions constitute all of the DM, eq. (10.31) implies a value of $f_a$ that depends on the initial value of the axion field, $a_* \equiv \theta_* f_a < \pi f_a$ (the upper bound arises because the axion field space forms a circle, $\theta = a/f_a < \pi$). Two main possibilities can be realized in cosmology:

1. If the PQ-breaking phase transition occurs before inflation, this is the so called '*pre-inflationary*' scenario, roughly corresponding to $f_a \gtrsim \max(H_{\mathrm{infl}}, T_{\mathrm{RH}})$, where $H_{\mathrm{infl}}$ is the inflationary Hubble scale and $T_{\mathrm{RH}}$ the reheating temperature. In this case $a_*$ is roughly constant within the horizon, and can be much smaller than $f_a$. The cosmological axion density matching the observed DM relic abundance can thus be reproduced even for large axion decay constants, $f_a \gg 10^{12}\,\mathrm{GeV}$, and thereby small axion masses [101]. However, axion inflationary fluctuation, $\delta a \sim H_{\mathrm{infl}}/2\pi$, lead to iso-curvature inhomogeneities

$$\left.\frac{\delta\rho_a}{\rho_{\mathrm{DM}}}\right|_{\mathrm{isoc}} = \frac{\rho_a}{\rho_{\mathrm{DM}}} 2\frac{\delta a_*}{a_*} \sim \frac{\rho_a}{\rho_{\mathrm{DM}}}\frac{H_{\mathrm{infl}}}{\pi f_a \theta_*}, \tag{10.32}$$

   which are constrained by the CMB data (measured at scales comparable to the horizon) to be less than about $\sim 20\%$ of the observed adiabatic fluctuations, $\delta\rho_{\mathrm{DM}}/\rho_{\mathrm{DM}} \sim 10^{-5}$. If all the DM is due to axions, $\rho_{\mathrm{DM}} = \rho_a$, this then implies a somewhat low inflationary Hubble scale, $H_{\mathrm{infl}} \lesssim 10^{-5}\theta_* f_a$, and thereby small tensor CMB modes. Observation of tensor CMB modes would exclud the 'pre-inflationary' scenario.

2. If PQ-breaking occurs in the thermal phase after inflation (the so called '*post-inflationary*' scenario, roughly corresponding to $f_a \lesssim \max(H_{\mathrm{infl}}, T_{\mathrm{RH}})$), then the thermal equilibrium is expected to wash out iso-curvature inhomogeneities. The axion field therefore acquires just the adiabatic inhomogeneities with the average $a_* \sim f_a$ [101]. In this case, the cosmological relic abundance is reproduced for [101]

$$m_a \approx 26\,\mu\mathrm{eV} \sim \frac{1}{0.75\,\mathrm{cm}}, \qquad \mathrm{i.e.} \qquad f_a \approx 2\;10^{11}\,\mathrm{GeV} \tag{10.33}$$

   if the axion production is dominated by the misalignment mechanism.

3. Intermediate possibilities arise, if $H_{\mathrm{infl}} \sim f_a$, where the exact range also depends on the couplings of the full model [101].

In the post-inflationary scenario (case 2) the PQ phase transition generates axion strings and possibly domain walls.

**Axion strings** give an extra contribution to the axion relic density that has not yet been reliably computed (computations are difficult because the energy density of axion strings is distributed logarithmically over distance scales from $1/f_a$ to the Hubble horizon). Such extra contribution could be sub-leading compared to the misalignment mechanism, but could even be up to 3 orders of magnitude larger. If this is the case, the axion mass suggested by cosmology could differ significantly from the estimate in eq. (10.33). Numerical simulations indicate an axion mass between 40 and 180 $\mu\mathrm{eV}$ [101].

**Domain walls** are absent if there is only one vacuum, $N_{\mathrm{DW}} = 1$. The number of non-equivalent vacua, $N_{\mathrm{DW}}$, is given by the QCD anomaly, i.e., by $\partial_\mu J^\mu_{\mathrm{PQ}} = \alpha_s N_{\mathrm{DW}} G\tilde{G}/4\pi + \cdots$, where $J^\mu_{\mathrm{PQ}}$ is the Noether current of the PQ symmetry. The value of $N_{\mathrm{DW}}$ thus depends on the UV theory. For $N_{\mathrm{DW}} > 1$ (as in DFSZ models) the domain walls are stable and carry enough energy to overclose the Universe, making such a theory of axions unviable. Additional ingredients are needed to make domain walls unstable, such as an explicit breaking of the PQ symmetry that lifts the degeneracy of the $N_{\mathrm{DW}}$ vacua, leading to domain annihilation. Even so, their extra contribution to the DM axion density seems so large that the DM cosmological abundance can only be reproduced for $f_a \lesssim$ few $10^8\,\mathrm{GeV}$, corresponding to $m_a \gtrsim 10\,\mathrm{meV}$, see Beyer and Sarkar in [101].

Furthermore, hot axions are produced thermally from gluon and quark scatterings, such as $gg \to ga$ with cross section $\sigma_a \approx \alpha_s^3/16\pi^2 f_a^2$. Such production of axions is dominated by the highest available temperature, which in this case is the reheating temperature $T_{\rm RH}$. It yields a thermal abundance of axions, $Y^a$,

$$\frac{Y_a}{Y_a^{\rm eq}} = \min(1, r), \qquad r = \frac{2.4}{Y_a^{\rm eq}} \left.\frac{\gamma_a}{Hs}\right|_{T=T_{\rm RH}} = 1.7\frac{T_{\rm RH}}{10^7\,{\rm GeV}}\left(\frac{10^{11}\,{\rm GeV}}{f_a}\right)^2 \left.\frac{\gamma_a}{T^6\zeta(3)/(2\pi)^5 f_a^2}\right|_{T=T_{\rm RH}}. \tag{10.34}$$

The last factor in eq. (10.34) is of order one, where the exact value depends on the precise value of the space-time density of the axion production rate, $\gamma_a \sim n_g^2 \sigma_a$. In the limit $r > 1$, i.e., in the limit of large enough interactions so that axions are fully thermalised, the hot axions take a form of extra background radiation with present day temperature of about $0.9\,^\circ$K. They would constitute a hot component of DM relic abundance, which is usually parameterised in terms of an "effective number of neutrinos", given by $\Delta N_{\rm eff} = 0.0264\,Y_a/Y_a^{\rm eq}$ [862], see eq. (C.27) in appendix C.3. Since $Y_a \le Y_a^{\rm eq}$ the corresponding $\Delta N_{\rm eff}$ is always well below the present constraints, $\Delta N_{\rm eff} \lesssim 0.3$.

Furthermore, since in the post-inflationary scenario the axion field takes random values $a_*$ in casually disconnected regions, this results in large, order unity inhomogeneities in the axion energy density around the time of QCD phase transition, $T \gtrsim \Lambda_{\rm QCD}$, when the axion mass becomes relevant. For sufficiently late times the mass term dominates and the axion energy density behaves like cold DM, with comoving number of axions conserved. The over-densities in axion energy distribution decouple from the Hubble expansion at $T \approx T_{\rm eq}$, leading to a gravitational collapse, forming **axion mini-clusters** [863]. Their mass is comparable to the mass within a Hubble volume at the time the axion field starts to oscillate, $M_{\rm Pl}^3/\Lambda_{\rm QCD}^2$. However, since the energy during this epoch is in radiation, it gets redshifted by a factor, $(T_{\rm eq}/\Lambda_{\rm QCD})$, giving the estimate $M_{\rm clust} \approx (T_{\rm eq}/\Lambda_{\rm QCD})M_{\rm Pl}^3/\Lambda_{\rm QCD}^2 \approx 10^{-11}M_\odot$. More compact objects, *axion stars*, can also form in the centers of axion DM halos. Axion stars will be discussed in section 10.5.4

Furthermore, some authors consider the possibility of **axion quark nuggets** [864]. Similarly to the quark matter discussed in section 9.1.2, these would be composed by many SM quarks, bound together in a non-hadronic high density phase, and would possess a large mass, $M \gtrsim 10^{24}\,{\rm GeV}$. Their formation, occurring around the QCD phase transition like for the quark matter, involves however the presence of the axion, whose domain walls would act as a stabilisation factor for the nugget, removing the need for a first order phase transition. Another difference is that the axion quark nuggets could be made of matter as well as antimatter (indeed the model addresses also the baryon asymmetry problem). They largely share the phenomenology of quark nuggets and can be tested in similar ways [864].

## 10.4.5 Searching for axion Dark Matter

In this section we discuss the phenomenology of the searches for axion and axion-like particles (ALPs). As already mentioned before (see the general discussion in section 3.4), the basic difference between the two is that for axions the mass and the couplings to the SM particles are related via the symmetry breaking scale $f_a$, while ALPs are the generalization of this concept, where the link between the mass and couplings is relaxed. For simplicity, we will use in the rest of this section the term axion in a broad sense, which includes ALPs.

Axions must couple to gluons, and are expected to also couple to the other SM fields. The axion Lagrangian of eq. (10.22) results at low energies, $\mu \lesssim$ few GeV, in the following couplings to gluons, photons, quarks and leptons,

$$\mathscr{L}_{\rm axion}^{\rm eff} = \mathscr{L}_{\rm QCD} + \frac{a}{f_a}\frac{\alpha_s}{8\pi}G_{\mu\nu}^a \tilde{G}^{a,\mu\nu} + c_\gamma \frac{a}{f_a}\frac{\alpha}{2\pi}F_{\mu\nu}\tilde{F}^{\mu\nu} + \frac{\partial_\mu a}{2f_a}\sum_{ij}\bar{\Psi}_i\gamma^\mu\left[C_{ij}^V + C_{ij}^A\gamma_5\right]\Psi_j, \tag{10.35}$$

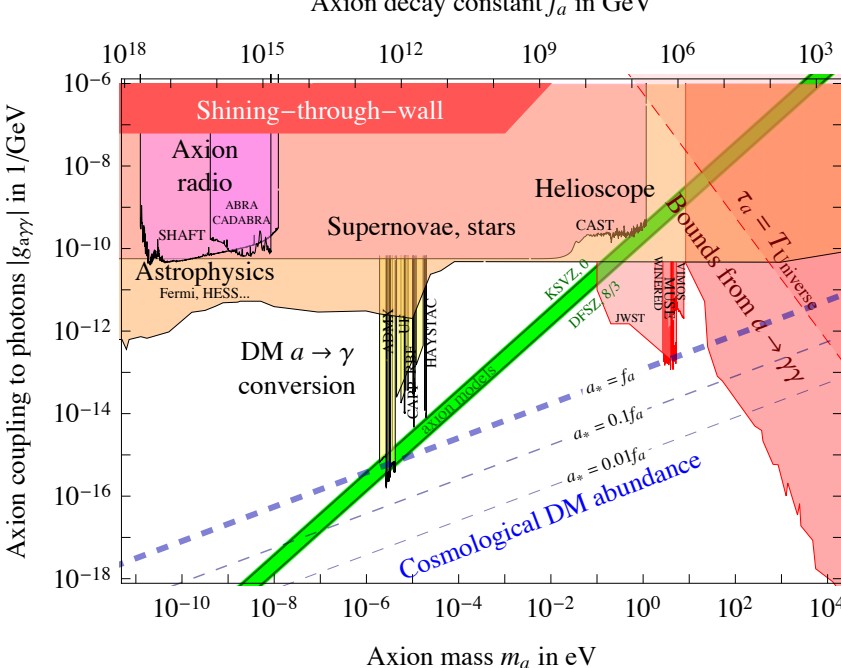

Figure 10.1: ***Axion and axion-like particle phenomenology.*** *In the plane (axion mass, axion coupling to photons) we show the range favoured by axion models (green band); the initial axion vacuum expectation value $a_*$ that reproduces the observed DM abundance (blue dashed lines); the exclusion bounds from non-observation of axion decays (upper-right red region) [865], as well as absence of signals in shining-trough-wall experiments (upper-left red region) [866–868], searches for axion emissions from the Sun (upper red region) [869, 870], stars, dwarfs and supernovæ (upper orange region) [871], and axion DM conversion into photons (yellow regions) [872–875]. The bounds are discussed in more details in the main text; additional and much refined bounds are collected at github.com/cajohare/AxionLimits (2024).*

where the sum is over three types of neutral currents, either over up-quarks, $\Psi_i = u, c$, down-quarks, $\Psi_i = d, s, b$, or leptons $\Psi_i = e, \mu, \tau$. Unlike in eq. (10.22) they are written in Dirac notation, and we kept the flavor-violating couplings, which can either result from integrating out the $W$ or are already present in the high-energy physics. Note that for $i = j$ the derivative couplings to vector currents do not contribute due to vector current conservation, so effectively one can set $C_{ii}^V = 0$.

**Searches relying on axion couplings to photons.** The axion almost inevitably couples to photons. If nothing else, the axion-gluon interaction induces at low energies the axion couplings to charged pions, and this in turn results in axion couplings to photons through loop corrections, see eq. (10.37) below. The axion coupling to photons, eq. (10.23), is the most important for axion phenomenology and is usually written as

$$\mathscr{L}_{a\gamma\gamma} = -\frac{g_{a\gamma\gamma}}{4} a F_{\mu\nu} \tilde{F}_{\mu\nu} = g_{a\gamma\gamma} a \boldsymbol{E} \cdot \boldsymbol{B}, \qquad g_{a\gamma\gamma} = \frac{\alpha c_\gamma}{2\pi f_a}, \qquad (10.36)$$

where $\tilde{F}_{\mu\nu} \equiv \frac{1}{2}\epsilon_{\mu\nu\alpha\beta}F_{\alpha\beta}$. The constant $g_{a\gamma\gamma}$ has dimension mass$^{-1}$ and $c_\gamma$ is a model-dependent order one number. Its explicit value, in the basis where the $aG\tilde{G}$ coupling is rotated away (as discussed below

eq. (10.24)) is

$$c_\gamma = c_{\gamma,\text{uv}} - \frac{2}{3}\frac{4 + m_u/m_d + m_u/m_s}{1 + m_u/m_d + m_u/m_s} \approx c_{\gamma,\text{uv}} - 1.93, \qquad \text{where} \qquad c_{\gamma,\text{uv}} = \frac{\sum_f X_f q_f^2}{\sum_f X_f T_f^2}. \tag{10.37}$$

The second term in $c_\gamma$ comes from the axion-gluon interaction, while $c_{\gamma,\text{uv}}$ is due to the triangle anomaly (with $a$ and two $\gamma$ as external legs) mediated by a loop of heavy fermions charged under both the global PQ symmetry and under the $\text{U}(1)_\text{em}$ gauge symmetry. The sums in $c_{\gamma,\text{uv}}$ are over Weyl fermions with PQ charges $X_f$ and electric charges $q_f$. The denominator appears because, by convention, $f_a$ is normalized relative to the QCD triangle anomaly (with $a$ and two $g$ as external legs). The QCD anomaly receives contributions only from colored fermions, where $\text{Tr}\, T^a T^b = T^2 \delta^{ab}$, with $T^a$ the SU(3) generators in the appropriate representation. For SU(3) triplets one has $T^2 = 1/2$, and thus DFSZ models, in which only SM fermions contribute to the anomaly, predict

$$c_{\gamma,\text{uv}} = \frac{N_c(q_d^2 + q_u^2) + q_e^2}{1/2 + 1/2} = \frac{8}{3} \tag{10.38}$$

where $N_c = 3$ is the number of colors in the SM, and we assumed that heavy fermions $f$ have common PQ charges $X_f$. For DFSZ models thus the two terms combine to $c_\gamma \simeq 0.73$, confirming the expectation that typically, $c_\gamma \sim 1$. KSVZ models too predict $E/N = 8/3$ whenever the extra heavy fermions fill multiplets under the SU(5) unification gauge group with common PQ charge.

Treating the background axion density as a classical field $a(x)$, see section 3.4, the inhomogeneous Maxwell equations get modified to

$$\boldsymbol{\nabla} \cdot \boldsymbol{E} = \rho - g_{a\gamma\gamma} \boldsymbol{B} \cdot \boldsymbol{\nabla} a, \qquad \boldsymbol{\nabla} \times \boldsymbol{B} = \boldsymbol{J} + \dot{\boldsymbol{E}} + g_{a\gamma\gamma}(\boldsymbol{B}\dot{a} - \boldsymbol{E} \times \boldsymbol{\nabla} a). \tag{10.39}$$

These re-acquire the Maxwell form if one defined the modified electric and magnetic fields, $\boldsymbol{E}' = \boldsymbol{E} + g_{a\gamma\gamma} a \boldsymbol{B}$, $\boldsymbol{B}' = \boldsymbol{B} - g_{a\gamma\gamma} a \boldsymbol{B}$. This shows that a varying axion field is able, in the presence of a background magnetic field, to produce electromagnetic effects such as electromagnetic waves, which can then be searched for.

Axions remain unobserved, implying that the scale $f_a$ suppressing the axion interactions with the SM fields, including $g_{a\gamma\gamma}$, see eq.s (10.35) and (10.36), must be very large. Fig. 10.1 summarizes in the $(m_a, g_{a\gamma\gamma})$ plane the rich axion phenomenology and the current bounds from many axion searches, while table 10.1 lists some of the main laboratory experiments leading to these bounds. Below, we provide a concise overview of the key aspects highlighted in fig. 10.1.

- The green band indicates the parameter space region predicted by the axion models according to eq.s (10.25) and (10.36). However, its width is somewhat arbitrary, and would change with more complicated field content, see, e.g., the discussion in [876].

- The blue dashed contour-lines show the initial axion vacuum expectation value $a_*$ that is required in order to reproduce the observed cosmological DM abundance via the misalignment mechanism, discussed in section 3.4, see also eq. (10.31) (to convert $f_a$ into $g_{a\gamma\gamma}$ we assumed $c_\gamma = 1$). The values $a_* \sim f_a$ (thick dashed line) are likely to be the more plausible ones.

The experimental constraints shown in fig. 10.1 are as follows.

- The $a\gamma\gamma$ interaction leads to the **axion decay** $a \to \gamma\gamma$. The corresponding lifetime

$$\tau(a \to \gamma\gamma) = \frac{64\pi}{g_{a\gamma\gamma}^2 m_a^3} \approx T_\text{U} \left(\frac{20\,\text{eV}}{m_a}\right)^5, \tag{10.40}$$

must be longer than the age of the universe, $T_U$. The most stringent bounds on the axion decay time are in fact much stronger, in the range $\tau > 10^{21-26}$ s, depending on the value of the axion mass $m_a$, and follow from the searches for an excess of such photons, and from bounds on the re-ionization of the Universe. These give the upper-right red excluded region in fig. 10.1 [865].

If the axion mass is around 1 eV, $a \to \gamma\gamma$ decays produce a flux of infrared or visible photons, which can be directly constrained by telescope observations such as Vimos, Muse, Winered, Jwst [865].

– Laboratory experiments searching for **light shining through wall** (photon–to–axion–to–photon conversion in a magnetic field) [866–868] presently set the weak bound in the top-left part of fig. 10.1. These searches will be improved by the upcoming experiment ALPS-II.

– The strong horizontal bound in the upper part of fig. 10.1, $g_{a\gamma\gamma} \lesssim 0.7\ 10^{-10}/\,\text{GeV}$, comes from the absence of excessive **cooling of stars** (including the Sun) and supernovæ, due to axion production. The axion cooling modifies the seismic properties of the star, its neutrino emission and in general its evolution. The former two effects, applied to the Sun, imply the bound $g_{a\gamma\gamma} \lesssim \text{few } 10^{-10}/\,\text{GeV}$. The more stringent constraint quoted above comes from analyzing the number ratio $R$ of stars in the horizontal branch (HB) over the red giant branch (RGB) in old galactic globular clusters. This ratio would be modified (reduced) by the axion production from photon conversions occurring in the stellar cores [871].

Notice also that the constraints from stellar and supernova cooling mentioned in section 7.4.1 apply [701, 702], typically to masses larger than those considered in fig. 10.1. Other stellar and supernova physics bounds on relatively heavy ALPs are considered in [877].

– The more stringent bound $g_{a\gamma\gamma} \lesssim \text{few } 10^{-12}/\,\text{GeV}$ at $m_a \lesssim 10^{-6}$ eV comes from the non-detection of irregularities in the spectra of a number of diverse astrophysical sources (quasars, blazars, clusters or individual galaxies) [878]. The photons emitted by these sources could **convert into axions** while crossing intergalactic magnetic fields. At a fundamental level this is the $\gamma\gamma \to a$ process, where the second $\gamma$ is the magnetic field. The conversion would dim the apparent intensity of the source, but this does not lead to testable effects given the uncertain randomness of the magnetic field and of the sources. It would also induce oscillatory features in the spectra of the sources, which are otherwise expected to be smooth, and this constitute a statistically testable signal. Bounds at different axion masses are derived from different $\gamma$- and $X$-ray experiments (e.g. Hess, Fermi, Magic, Chandra) [878].

Still in the vein of conversion, other bounds can be derived from the non-observation of excess $X$-rays or radio-waves from relatively $X$-dim or radio-quiet sources [879]. If these sources emit axions, these could **convert into photons** in the magnetic fields along the way. At particle level, the process is now $a\gamma \to \gamma$, where the $\gamma$ in the initial state is the magnetic field. This is particularly efficient in the intense magnetic fields surrounding neutron stars. The non-observation of excess radio emission from pulsars (rotating neutron stars), or excess $X$-ray emission from Betelgeuse or some white dwarfs, or even of an excess in the $\gamma$-ray background from past supernovæ allows to impose $g_{a\gamma\gamma} \lesssim \text{few } 10^{-11-12}/\,\text{GeV}$ in various mass ranges, comparable to the stellar cooling bounds [879].

A hint of a positive signal due to axion-photon conversion in some isolated neutron stars (known as the *Magnificent Seven*) is however claimed by Bushmann et al. (2019) in [879].

– While the Sun is not the hottest star, its proximity allows for **helioscope** experiments [880], which use intense magnetic fields to search for conversions of axions, emitted by the Sun, into photons (again $a\gamma \to \gamma$, where now the $\gamma$ in the initial state is the strong magnetic field in the experiment). Helioscopes (such as CAST [869]) set bounds similar to the stellar cooling ones for

$m_a \lesssim 0.02\,\text{eV}$. Future improvements are planned: the International Axion Observatory (IAXO) should improve the CAST reach on $f_a$ by more than an order of magnitude, surpassing the stellar cooling bounds [870].

Observations of the Sun can also be used to impose constraints on axions in a different mass range, $m_a \approx 10\,\text{keV}$. For these masses, a significant fraction of axions thermally produced in the solar core would possess a speed below the escape velocity and would remain gravitationally bound to the Sun, thereby accumulating in a **solar axion basin**, eventually decaying into $X$-rays outside the Sun [881]. $X$-ray observations imply the constraint $g_{a\gamma\gamma} \lesssim \text{few } 10^{-13}/\text{GeV}$ (not shown in fig. 10.1).

– **Haloscope** experiments [880] use the $a\gamma\gamma$ coupling to search for resonant conversion of galactic axion DM into a microwave photon with energy $E_\gamma = m_a$ using a microwave cavity of volume $V$, permeated by a strong magnetic field $B_0$. The power generated within the cavity by axion to photon conversion is given by

$$W = g_{a\gamma\gamma}^2 V B_0^2 \frac{\rho_\odot}{m_a} C \min(Q_{\text{cavity}}, Q_{\text{axion}}), \qquad (10.41)$$

where $C$ is a geometric factor (e.g., $C = 0.69$ for the $\text{TM}_{010}$ mode of a cylindrical cavity permeated by a longitudinal $B_0$), $Q_{\text{cavity}}$ is the quality factor of the cavity and the inverse of $Q_{\text{axion}}$ measures the spread in the axion energy $E_a \simeq m_a c^2 + \frac{1}{2} m_a v^2$, so that $Q_{\text{axion}} \approx (c/v)^2 \approx 10^6$. Streams of axions with common velocity would all have the same energy, which would then enhance the signal-to-noise ratio.

By varying the resonant frequency of the cavity one can probe different axion masses (yellow dips in fig. 10.1). These searches already reach the sensitivity to the parameter space predicted by the most popular axion models, but only in narrow ranges of axion masses, because relatively long running times are needed at each frequency in order to accumulate sufficient signal. Recent and current haloscopes include RBF, UF, ADMX, HAYSTAC, QUAX-$a\gamma$, CAPP [872–875, 882], as well as many others (see table 10.1).

There are also a number of other experimental approaches, some of which target axion masses that are not in the ranges considered to be the most plausible (see [858] for further details):

– **Dish antennae and dielectric haloscopes**. If an external magnetic field is aligned along a conducting surface, such as a mirror, or a surface of a block of dielectric material, the oscillating DM axion field excites the electrons inside the material, leading to the emission of electromagnetic radiation from the surface of the material. Future searches for the electromagnetic radiation emitted by large surface magnetised dish antennae and dielectric haloscopes, such as MADMAX, consisting of a large array of dielectric slabs, are expected to reach considerable sensitivity [883].

– **Axion DM radios**. The oscillating DM axion field in a constant external magnetic field induces a small oscillating $\boldsymbol{B}$ field. This can then be detected using pick-up coils inside a larger magnet. Table top size experiments based on this idea (e.g., ABRACADABRA) placed limits comparable to those from the CAST helioscope, down to axions masses $m_a \sim 10^{-8-10}\,\text{eV}$. This technique could reach sensitivities around the QCD axion band in fig. 10.1 for axions lighter than about $1\,\mu\text{eV}$, if magnets with $B \sim \text{few T}$ and of cubic meter volumes can be used in the future [884–888]. A similar technique, using the whole Earth as the test volume, has been used to impose constraints on very light axions or dark photons ($m_a \sim 10^{-17}\,\text{eV}$), using data from the SUPERMAG Collaboration [889].

– **Polarization experiments.** Photon-axion conversions in the presence of an external magnetic field $\boldsymbol{B}$ would reduce the polarization component of a laser beam parallel to $\boldsymbol{B}$ (dichroism) and

delay its phase (birefringence). This effect has been searched for, e.g., by PVLAS [890], obtaining relatively weak bounds, $|g_{a\gamma\gamma}| \lesssim 10^{-7}/\,\mathrm{GeV}$.

**Searches relying on couplings to gluons and/or SM fermions.**    While most axion searches focus on couplings to photons, the axion couplings to gluons or SM fermions, see eq. (10.35), can lead to additional signals [5]. Tables 10.2 and 10.3 list the main experiments probing these other interactions. See also a similar, but more general, discussion in section 5.8.

- The axion couplings to gluons and quarks generate, at low energies, a coupling between the axion field and the nucleon spins. The most relevant operators are

$$\mathscr{L}_a^{\mathrm{eff}} = g_{aN}(\partial_\mu a)\bar{N}\gamma^\mu\gamma_5 N - \frac{i}{2}g_{ad}\,a\bar{N}\sigma_{\mu\nu}\gamma_5 N F^{\mu\nu}. \tag{10.42}$$

As discussed above, the local DM axion field oscillates with frequency determined by its energy $E_a \simeq m_a$, so that the first interaction sources an **oscillating magnetic dipole** with coupling $g_{aN} \propto 1/f_a$, and the second an **oscillating electric dipole**, with $g_{ad} \propto 1/m_N f_a$. For nonrelativistic nucleons, which is the case for nucleons bound inside a nucleus, the leading non-relativistic Lagrangian following from eq. (10.42) is given by

$$\mathscr{L}_a^{\mathrm{NR}} = -\gamma_N \boldsymbol{B}_a^{\mathrm{eff}} \cdot \boldsymbol{S}_N, \qquad \boldsymbol{B}_a^{\mathrm{eff}} = -\frac{2}{\gamma_N}[g_{aN}\boldsymbol{\nabla}a + g_{ad}a\boldsymbol{E}]. \tag{10.43}$$

Here, $\boldsymbol{S}_N$ is the spin of the nucleon, and $\boldsymbol{E}$ the electric field acting on the nucleon (for nucleons bound inside nucleus this differs from the applied external electric field due to shielding by the electronic cloud). The oscillating axion therefore plays the role of an effective magnetic field $B_a^{\mathrm{eff}}$, where $\gamma_N$ is the gyromagnetic ratio of the nucleon. Such an oscillating axio-magnetic field can be detected via induced precession of nuclear spins, and can, in the future, be searched for via nuclear magnetic resonance techniques [418]. These experiments may in the long term reach the sensitivity to the nominal QCD axion band (the green band in figure 10.1), if $m_a \sim \mu\mathrm{eV}$.

- For flavor diagonal couplings to quarks, **colliders** set bounds on axions (and more generic axion-like particles) that have relatively small decay constants, $f_a \sim \mathrm{TeV}$, and therefore large masses, $m_a$, see sections 7 and 9.4.4.

- **Production in flavor violating decays**. Flavor violating axion couplings induce flavor-changing neutral current decays of quarks, such as $s \to da$ and $b \to sa$, and of leptons, $\mu \to ea$ and $\tau \to ea, \mu a$ [927]. One can consider two limits: large and small flavor violating couplings. In the first case, i.e., if the flavor violating couplings are present in the full theory then generically $C_{ij}^{A,V} \approx 1$ and such decays give stringent bounds, typically above $f_a \gtrsim 10^9$ GeV. In this case the QCD axion is light, $m_a \ll \mathrm{eV}$ (see eq. (10.25)), and has a cosmologically long lifetime (see eq. (10.40)). Decays such as $K \to \pi a$ and $B \to Ka$ therefore appear in the experiments as decays with missing energy, since the axions escape the detector before decaying. The resulting bounds are comparable or even more stringent than the astrophysical constraints and the bounds from terrestrial searches for axions via their coupling photons [928]. The opposite limit to large flavor violating couplings is that the axion couplings are flavor diagonal at tree level, while the flavor violating couplings are only generated at one loop from the SM CKM structure. This is, for instance, the case in the original DFSZ and KSVZ models. In that case, the bounds on $f_a$ are significantly weaker, at the TeV scale, and thus the axion can be heavier, in the keV regime (if

---

[5]The coupling to fermions can in turn also induce a two-photon coupling, via a fermionic triangle loop, see e.g. [926].

| Experiment | Location | Operation | Technology | Mass Range | | | Coupling sensitivity (GeV$^{-1}$) | Home | Ref. |
|---|---|---|---|---|---|---|---|---|---|
| SuperMAG | Worldwide | 1970 $\to$ | Magnetic Field HS | 2 | $\to$ | 70 aeV | 6.5 $10^{-11}$ | web | [889] |
| RBF | BNL, USA | 1987 | $\mu$-wave Cavity HS | 4.5 | $\to$ | 5.0 $\mu$eV | $10^{-11}$ | – | [882] |
| UF | UoFlorida, USA | 1989 | $\mu$-wave Cavity HS | 5.4 | $\to$ | 5.9 $\mu$eV | 3.7 $10^{-14}$ | – | [891] |
| ADMX { | LLNL, USA | 1995 $\to$ 2010 | $\mu$-wave Cavity HS | 1.9 | $\to$ | 3.69 $\mu$eV | 1 $10^{-15}$ | web | [872] |
| | CENPA, USA | 2017 $\to$ | | 2.66 | $\to$ | 4.2 $\mu$eV | 3.5 $10^{-16}$ | | |
| CAST { | CERN | 2004 | Helioscope | $\lesssim$ | | 20 meV | 8.8 $10^{-11}$ | web | [869] |
| | | 2013 $\to$ 2015 | | $\lesssim$ | | 20 meV | 6.6 $10^{-11}$ | | |
| ALPS { | DESY, Germany | 2007 $\to$ 2010 | LSW | $\lesssim$ | | 1 meV | 7 $10^{-8}$ | web | [866] |
| | | 2023? | | $\lesssim$ | | 1 meV | 2 $10^{-11}$ | | |
| Crows | CERN | 2013 | Microwave LSW | $\lesssim$ | | 7.2 $\mu$eV | 4.5 $10^{-8}$ | – | [868] |
| Osqar | CERN | 2014 | LSW | $\lesssim$ | | 0.2 meV | 3.5 $10^{-8}$ | web | [867] |
| Pvlas | Padova, Italy | 2015 | Optical Birefringence | $\lesssim$ | | 0.01 eV | 1 $10^{-7}$ | wiki | [890] |
| ADMX Sidecar | CENPA, USA | 2016 $\to$ | $\mu$-wave Cavity HS | 17.38 | $\to$ | 29.79 $\mu$eV | 2.6 $10^{-13}$ | web | [872] |
| Haystac | Yale, USA | 2016 $\to$ 2021 | $\mu$-wave Cavity HS | 16.96 | $\to$ | 24.0 $\mu$eV | 9.07 $10^{-15}$ | web | [873] |
| Organ | UWA, Australia | 2017 $\to$ 2022 | $\mu$-wave Cavity HS | 63.2 | $\to$ | 110 $\mu$eV | 2.02 $10^{-12}$ | – | [892] |
| Abracadabra | MIT, USA | 2018 | Lumped Element HS | 0.31 | $\to$ | 8.3 neV | 3.2 $10^{-11}$ | web | [885] |
| ADMX SLIC | UoFlorida, USA | 2018 | Lumped Element HS | 0.175 | $\to$ | 0.180 $\mu$eV | $10^{-12}$ | web | [872] |
| QUAX-$a\gamma$ | LNL, Italy | 2018 $\to$ 2021 | $\mu$-wave Cavity HS | 37.5 | $\to$ | 42.8 $\mu$eV | 7.31 $10^{-14}$ | web | [874] |
| RADES | CERN | 2018 | RF Cavity HS | $\simeq$ | | 34.67 $\mu$eV | 4 $10^{-13}$ | web | [893] |
| CAPP | CAPP, S. Korea | 2019 $\to$ | $\mu$-wave Cavity HS | 4.5 | $\to$ | 19.9 $\mu$eV | $\sim 10^{-15}$ | web | [875] |
| CAST-CAPP | CERN | 2019 $\to$ 2021 | $\mu$-wave Cavity HS | 19.74 | $\to$ | 22.47 $\mu$eV | 8 $10^{-14}$ | web | [894] |
| SHAFT | Boston Univ. | 2019 | Lumped Element HS | 12 | $\to$ | 12 $10^3$ peV | 4 $10^{-11}$ | – | [886] |
| UPLOAD { | UWA, Australia | 2019$^\dagger$ | $\mu$-wave Cavity HS | 7.44 | $\to$ | 19.38 neV | 3 $10^{-13}$ | – | [895] |
| | | 2023$^\dagger$ | | 1.12 | $\to$ | 1.20 $\mu$eV | 3 $10^{-6}$ | | |
| BASE | CERN | 2020$^\dagger$ | Penning Trap HS | $\simeq$ | | 2.79 neV | $10^{-11}$ | web | [887] |
| DANCE { | UoTokyo, Japan | 2021 | Optical Cavity HS | $10^{-14}$ | $\to$ | $10^{-13}$ eV | 8 $10^{-4}$ | slides | [896] |
| | | Proposed | | $\lesssim$ | | $10^{-10}$ eV | 3 $10^{-16}$ | | |
| GrAHal { | Grenoble, France | 2021 | $\mu$-wave Cavity HS | $\simeq$ | | 26.37 $\mu$eV | 2.2 $10^{-13}$ | web | [897] |
| | | 2023? | | 1.25 | $\to$ | 125 $\mu$eV | 1.1 $10^{-14}$ | | |
| Lampost | MIT, US | 2021 | Optical Dielectric HS | 0.1 | $\to$ | 10 eV | $\sim 10^{-12}$ | – | [232] |
| Taseh | NCU, Taiwan | 2021 | $\mu$-wave Cavity HS | $\simeq$ | | 19 $\mu$eV | 8.2 $10^{-14}$ | web | [898] |
| Sapphires | Kyoto Univ., Japan | 2022$^\dagger$ | Laser Collider | 0.005 | $\to$ | 0.5 eV | 1.14 $10^{-5}$ | web | [899] |
| WISPLC | UHH, Germany | 2023? | Lumped Element HS | $10^{-11}$ | $\to$ | $10^{-6}$ eV | $10^{-15}$ | web | [900] |
| Baby-IAXO | DESY, Germany | 2024? | Helioscope | $\lesssim$ | | 0.25 eV | 1.5 $10^{-11}$ | web | [870] |
| IAXO | DESY, Germany | 2030? | Helioscope | $\lesssim$ | | 1 eV | $10^{-12}$ | web | [870] |
| Madmax | DESY, Germany | 2030? | Dielectric HS | 40 | $\to$ | 400 $\mu$eV | $10^{-14}$ | web | [883] |
| Bread | Fermilab, USA | Pilot | Dish Antenna HS | $10^{-4}$ | $\to$ | 1 eV | $\sim 10^{-13}$ | – | [901] |
| CADEx | LSC, Spain | Prep. | $\mu$-wave Cavity HS | 330 | $\to$ | 460 $\mu$eV | 3 $10^{-13}$ | slides | [902] |
| WISPfi | UHH, Germany | Constr. | Fiber Interferometer | 50 | $\to$ | 100 meV | $10^{-11}$ | web | [903] |
| DMRadio | SLAC, USA | Constr. | Lumped Element HS | 0.02 | $\to$ | 830 neV | $10^{-17}$ | web | [888] |
| aLIGO | LIGO, USA | Proposed | Optical Cavity HS | $10^{-16}$ | $\to$ | $10^{-9}$ eV | 8.3 $10^{-13}$ | web | [904] |
| ALPHA | Stockholm Univ., Sweden | Proposed | Plasma HS | 35 | $\to$ | 400 $\mu$eV | $\sim 10^{-14}$ | web | [905] |
| BRASS | Hamburg, Germany | Proposed | Broadband HS | 10 | $\to$ | $10^4$ $\mu$eV | – | web | – |
| Dali | Teide Observatory, Spain | Proposed | Dielectric HS | 25 | $\to$ | 250 $\mu$eV | 3.75 $10^{-15}$ | web | [906] |
| Flash | LNF, Italy | Proposed | $\mu$-wave Cavity HS | 0.2 | $\to$ | 1 $\mu$eV | 5.3 $10^{-15}$ | web | [907] |
| SRF/Dark-SRF | Fermilab, USA? | Proposed | Supercond. RF HS | $10^{-22}$ | $\to$ | $10^{-7}$ eV | $10^{-18}$ | web | [908] |
| Toorad | Padova, Italy | Proposed | Topol. Magn. Ins. HS | 1 | $\to$ | 10 meV | 4 $10^{-12\,\ddagger}$ | article | [909] |

Table 10.1: *Main **axion search experiments**, sensitive to the **axion-photon coupling** $g_{a\gamma\gamma}$, eq. (10.36). In the table, HS designates the* haloscopes, *while LSW designates the* light shining through a wall *experiments. The symbol* $^\dagger$ *indicates initial publication year in case the run period of experiment is not mentioned. The symbol* $^\ddagger$ *indicates that the sensitivity depends on the choice of the material. We gratefully acknowledge the work of Aryaman Bhutani for the content of this table.*

| Experiment | Location | Operation | Technology | Mass Range | | Coupling sensitivity (GeV$^{-1}$) | Home | Ref. |
|---|---|---|---|---|---|---|---|---|
| NEDM | Grenoble, France | 1998 → 2016 | neutron EDM | $10^{-24}$ → | $10^{-17}$ eV | $\sim 4\ 10^{-5}$ | article | [910] |
| SNO | Sudbury, Canada | 1999 → 2006 | neutrino obs | $\lesssim$ | 5.5 MeV | $1.88\ 10^{-15}$ | web | [911] |
| K-$^3$He Comagnetometer | Princeton, USA | 2008 | Comagnetometer | 0.04 → | 40 feV | $2.4\ 10^{-10}$ | – | [912] |
| PSI HgM | PSI, Switzerland | 2017 | Spin Precession | $10^{-16}$ → | $10^{-13}$ eV | $3.5\ 10^{-6}$ | web | [913] |
| CASPEr-ZULF | Mainz, Germany | 2018 | NMR/Comagnetometer | $10^{-22}$ → | $7.8\ 10^{-14}$ eV | $6\times 10^{-5}$ | web | [914] |
| JEDI | Jülich, Germany | 2019 | Storage Ring | 0.495 → | 0.502 neV | $1.5\ 10^{-5}$ | web | [915] |
| Hefei Xe129 | Hefei, China | 2021† | Quantum Sensor | 8.3 → | 744 feV | $5.44\ 10^{-9}$ | – | [916] |
| NASDUCK Floquet | Haifa, Israel | 2021 | Atomic Spin | $4\ 10^{-15}$ → | $4\ 10^{-12}$ eV | $1\ 10^{-7}$ | – | [917] |
| NASDUCK SERF | Haifa, Israel | 2022 | Atomic Spin | $1.4\ 10^{-12}$ → | $2\ 10^{-10}$ eV | $5\ 10^{-6}$ | – | [918] |
| ChangE | Beijing, China | 2023 | Quantum Sensor | $4\ 10^{-17}$ → | $4\ 10^{-12}$ eV | $3\ 10^{-10}$ | – | [919] |
| CASPEr | Mainz, Germany | Projection | NMR | $1\ 10^{-14}$ → | $1\ 10^{-6}$ eV | $\sim 1\ 10^{-14}$ | web | [920] |
| Superfluid He-3 HPD | – | Projection | NMR | 7 → | 74 neV | $\sim 1\ 10^{-12}$ | – | [921] |

Table 10.2: *Main **axion search experiments**, sensitive to the dimensionless **axion-nucleon coupling** $g_{aN}$. In the table, HS designates a* haloscope. *The symbol* $^\dagger$ *indicates initial publication year in case the run period of experiment is not mentioned. We gratefully acknowledge the work of Aryaman Bhutani for the content of this table.*

| Experiment | Location | Operation | Technology | Mass Range | | | Coupling sensitivity | Home | Ref. |
|---|---|---|---|---|---|---|---|---|---|
| DarkSide-50 | LNGS, Italy | 2013 → 2020 | LAr | 30 → | 200 | eV | $5\ 10^{-13}$ | web | [325] |
| | | | | 30 → | 20000 | eV | $1\ 10^{-12}$ | | |
| LUX | Sanford, SD | 2013 | LXe | 1 → | 16 | keV | $4.2\ 10^{-13}$ | web | [326] |
| EDELWEISS-III | LSM, France | 2014 → 2015 | Ge detector | 0.8 → | 500 | keV | $1\ 10^{-11}$ | web | [329] |
| PandaX-II(solar) | Jinping, China | 2014 → 2019 | LXe | $10^{-2}$ → | $10^3$ | eV | $4.35\ 10^{-12}$ | web | [330] |
| PandaX-II(galactic) | | | | 1 → | 25 | keV | $4.3\ 10^{-14}$ | | |
| SuperCDMS | Soudan, MN | 2014 → 2015 | Ge detector | 0.04 → | 500 | keV | $\sim 10^{-12}$ | web | [322] |
| GERDA | LNGS, Italy | 2015 → 2018 | Ge detector | 0.06 → | 1 | MeV | $3\ 10^{-12}$ | web | [922] |
| XENON1T | LNGS, Italy | 2016 → 2018 | LXe | 0.186 → | 400 | keV | $5\ 10^{-15}$ | web | [185] |
| QUAX-*ae* | LNL, Italy | 2018 → 2019 | YIG ferromagn. HS | 4.4 → | 58 | μeV | $1.7\ 10^{-11}$ | web | [923] |
| XENONnT | LNGS, Italy | 2021 | LXe | 1 → | 140 | keV | $\sim 10^{-14}$ | web | [727] |
| Magnon QND | UoTokyo, Japan | 2021† | magnon excitation | ≃ | 33.1 | μeV | $2.6\ 10^{-6}$ | – | [924] |
| LZ | Sanford, SD | 2022 → | LXe | 2 → | 20 | keV | $1.5\ 10^{-13}$ | web | [925] |

Table 10.3: *Main **axion search experiments**, sensitive to the dimensionless **axion-electron coupling** $g_{ae}$. In the table, HS designates a* haloscope. *The symbol* $^\dagger$ *indicates initial publication year in case the run period of experiment is not mentioned. We gratefully acknowledge the work of Aryaman Bhutani for the content of this table.*

there are additional contributions to its mass beyond the SM QCD anomaly, the mass can even be large enough that the axion decays inside the detector, completely changing the phenomenology of searches).

■ Axions can **couple to electrons**, in which case they induce excitations or ionizations of the target atoms, producing the same signals as discussed in the context of direct DM detection searches in section 5.2. Hence, a number of **direct detection experiments** have been used to probe the axion parameters, see table 10.3. Conventionally, the Lagrangian axion/electron interaction is written in terms of the rescaled dimensionless coupling to electrons $g_{ae}$ as

$$g_{ae}\frac{\partial_\mu a}{2m_e}\bar{e}\gamma^\mu\gamma^5 e \simeq -g_{ae}a\,\bar{e}i\gamma^5 e \qquad \text{where} \qquad g_{ae} \equiv -\frac{m_e}{f_a}C_{11}^A \qquad (10.44)$$

with $C_{11}^A$ given in eq. (10.35). Similar to the case of axion-photon coupling discussed above, the axion coupling to electrons also has an impact on the **cooling of stars**. A variety of axion production processes can occur in stellar cores due to this coupling, including $A$tomic axio-recombination and axion-deexcitation, axio-$B$remsstrahlung in electron-ion or electron-electron collisions and $C$ompton scattering with the emission of an axion, collectively known as the $ABC$ processes. The absence of excessive cooling, in particular in red giants, imposes a constraint $g_{ae} < 1.3\,10^{-13}$ [929].

Recently, however, a hint of faster-than-expected cooling in white dwarfs emerged, using several different techniques. This can be interpreted as being due to an axion with $g_{ae} \sim 1.5\,10^{-13}$ [929].

**Other constraints.**   At the low end of the mass spectrum, i.e., for very light axions ($m_a \lesssim 10^{-11}$ eV), one can also search for their gravitational effects:

– **Super-radiance**. If the radii of stellar black holes are comparable to the axion wave-length the black holes would become surrounded by axion clouds (see also section 6.14). Such a gravitationally bound state would extract the angular momentum of the black hole, if the axion self-coupling is sufficiently small (a requirement typically satisfied for $f_a \gtrsim 10^{18}$ GeV). The observations of black hole angular momenta imply $f_a \lesssim 0.6\,10^{18}$ GeV or $f_a \gtrsim M_{\rm Pl}$ [930]. Additional constraints are obtained for very light axions, $m_a \lesssim 10^{-17}$ eV, using polarimetric observations of supermassive black holes via Event Horizon Telescope, if axions have a relatively large coupling to photons, corresponding to an axion decay constant of $f_a \lesssim 10^{16}$ GeV [930].

– **Mass-radius relation for stellar remnants**. In large and dense systems such as white dwarfs the large number density of axions can reduce the in-medium neutron mass by tens of MeV, due to the $aG\widetilde{G}$ axion coupling to gluons, see eq. (10.35). This would modify the mass-radius relation for such stellar remnants. The observations of white dwarfs instead agree with the SM expectations, and thus exclude $f_a \lesssim 10^{16}$ GeV for very light axions, $m_a \lesssim 10^{-13}$ eV, and $f_a \lesssim 10^{16}$ GeV $(10^{13}$ eV$/m_a)$ for axions heavier than $10^{-13}$ eV [931] (the change in the scaling occurs for $1/m_a$ comparable to the white dwarf size). These constraints are not stringent enough to probe the QCD axion.

A more in depth review of searches and bounds on axion dark matter can be found, e.g., in O'Hare (2024) in [1].

## 10.5   Dark Matter as macroscopic objects

Section 3.2 discussed the possible signals of DM, if this consists of macroscopic objects heavier than the Planck scale [99]. In this section, we discuss the possible theories that can result in such DM, as well as the cosmological formation mechanisms for macroscopic objects, focusing on cases other than the primordial black holes, which were already discussed in section 4.6.1.

Before we start, it is important to note that ordinary matter is not a trivial phenomenon from the perspective of fundamental particle physics. First of all, ordinary liquids or solids are possible because there exist two stable particles, the electron and the proton, which carry opposite charges under the long-range U(1)$_{\rm em}$ gauge interaction. Ordinary matter has a constant typical density $\rho \sim M/R^3 \sim \alpha^3 m_p m_e^3$ [81] (see also section 3), because the mass $M$ and volume of macroscopic objects are proportional to those of constituents (atoms with nuclear mass $\sim m_p$, and Borh radius $\sim 1/\alpha m_e$). A large number of atoms can form macroscopic objects thanks to the fact that the electron and proton abundances are asymmetric: they carry conserved baryon and lepton numbers, and they did not annihilate away completely during cosmological evolution. Furthermore, the existence of hydrogen requires that the lightest state with nonzero baryon number is charged, i.e., that a proton is lighter than a neutron. In the SM this is the case even though the proton, as a charged particle, has a higher electromagnetic self energy and thus the

electromagnetic contribution to the mass than the neutron. The reason is that the $u$ quark is sufficiently lighter than the $d$ quark, which acts in the opposite direction from the electromagnetic effects. Chemistry more complex than the one for just the hydrogen atoms exists because neutrons too become stable within nuclei: the nuclear binding energy is large enough to kinematically block neutron decays. The atoms can then build large structures with arbitrary size because of the attractive electromagnetic dipole interaction acting even among neutral atoms. This can be compared to nuclear matter, that only exists in the form of small nuclei, up to $\sim 100$ protons and neutrons, or in the form of enormous, gravitationally bound, neutron stars, because neutrons repel each, see eq. (9.4).

Which brings us to the topic of 'dark matter'. Ironically, despite its name, DM theories in which dark particles actually form macroscopic solid objects analogous to ordinary matter, have mostly not been studied in any great detail. The reason is that this phenomenon often requires a rather complex dark sector with dark forces, complicating the analysis.

## 10.5.1   Macroscopic dark objects made out of heavy free particles

A minimal possibility for a DM theory that results in macroscopic dark objects is a theory of nearly-free and stable fermions or bosons with mass $m$; if their self-interactions are small enough, the gravitational attractive force leads to a formation of macroscopic objects that is compatible with the 'gravity as the weakest force' conjecture [855]. We start with the fermionic case, and then discuss the bosonic case, in both cases assuming that any interactions beyond gravitational force can be ignored.

Let us consider a Dirac *fermion* with $N$ components and mass $m$, and for simplicity assume that we can work in the non-relativistic gravity limit. A sphere of radius $R$ containing $Q \gg 1$ such fermions, and of uniform density, has the energy

$$U = U_{\text{quantum}} + U_{\text{grav}}, \qquad U_{\text{quantum}} = \frac{9}{20} \left( \frac{3\pi^2}{2N^2} \right)^{1/3} \frac{Q^p}{mR^2}, \qquad U_{\text{grav}} = -\frac{3Q^2m^2}{5RM_{\text{Pl}}^2}, \tag{10.45}$$

where the exponent in $U_{\text{quantum}}$ is $p = 5/3$. The energy is minimal when $\wp_{\text{quantum}} + \wp_{\text{grav}} = 0$, where $\wp_i = \partial U_i / \partial V$ are the corresponding pressure terms. Ignoring order one factors, the above equations can be understood as $\wp_{\text{grav}} \sim -U_{\text{grav}}/R^3 \sim Q^2 m^2/R^4 M_{\text{Pl}}^2$ and $\wp_{\text{quantum}} \sim -nK \sim -n^{5/3}/m \sim -Q^{5/3}/mR^5$, where $n \sim Q/R^3$ is the number density. That is, the Fermi pressure $\wp_{\text{quantum}}$ arises because fermions fill energy levels up to the Fermi momentum $k \sim n^{1/3}$, which in the non-relativistic limit corresponds to the kinetic energy $K = k^2/2m$. The total energy $U$ is minimal at $R \sim M_{\text{Pl}}^2/m^3 Q^{1/3}$ up to $Q \lesssim (M_{\text{Pl}}/m)^3$. Above this value the system becomes relativistic so that $\wp_{\text{quantum}} \sim n^{4/3} \sim Q^{4/3}/R^4$ can no longer compensate $\wp_{\text{gravity}}$, as now the two pressure terms have the same $R$ dependence. Around the same value of $Q$ also gravity becomes strong and a black hole starts to form. That is, these **gravitational Fermi balls** with masses $M \sim Qm$ thus have the radius to mass relation, $R \sim M_{\text{Pl}}^2/m^{8/3} M^{1/3} \propto M^{-1/3}$, that is quite different from the ordinary matter. It would correspond to a new DM line in the 'Catalogue of the Universe', fig. 3.1, with a downward slope (contrary to positive slopes for ordinary matter). The line would terminate at the black hole mass $\sim M_{\text{Pl}}^3/m^2$, and therefore the intercept of the gravitational Fermi ball line with the black hole boundary would tell us what the particle mass $m$ is. Since the strength of gravitational interaction is dimension-full (suppressed by the Planck mass) and becomes weaker for a smaller number of particles $Q$, the density $\rho \sim M/R^3$ of this form of matter is not constant (unlike for normal matter), but rather grows with the number of fermions as $\rho \propto Q^2$.

Let us next consider the case of stable *bosons* with mass $m$. The energy differs from the fermionic case of eq. (10.45): $U_{\text{quantum}}$ is now lower

$$U_{\text{quantum}} \sim QK \sim Q^p/mR^2, \qquad \text{with} \quad p = 1, \tag{10.46}$$

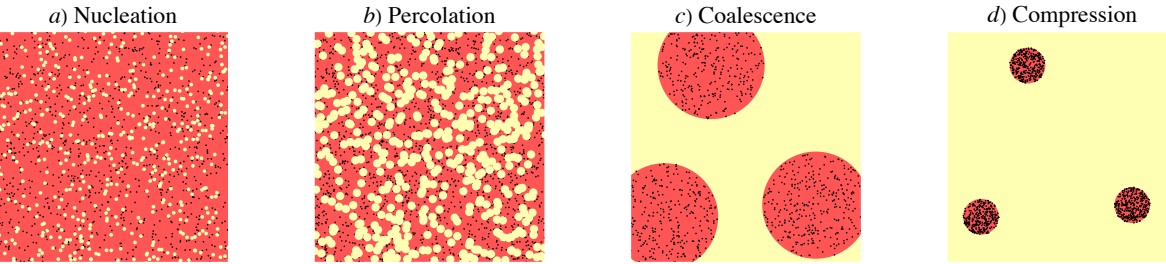

Figure 10.2: *Cartoon of how a first order phase transition can lead to the compression of DM particles into dense bodies. Bubbles of the new phase (in yellow) appear and expand. DM particles, denoted as black dots, cannot enter the new phase and thereby get compressed into remaining small pockets that form after the bubbles merge.*

since all modes can have the minimal $k \sim 1/R$ and thus the kinetic energy per particle is $K = k^2/2m \sim 1/mR^2$. While the fermion pressure is an intensive quantity that can be written in terms of the number density $n$, a non-relativistic boson gas instead has a pressure $\wp_{\text{quantum}} \sim Q/mR^5 \sim n/mR^2$ that depends on the size of the system. For macroscopic $R$ this Casimir-like effect can easily be a sub-dominant contribution to the energy relative to the potential energy due to scalar self-interactions. Ignoring self-interactions, the **gravitational Bose balls** would have a smaller radius and bigger mass for given $Q$ than gravitational Fermi balls: $R \sim M_{\text{Pl}}^2/m^3Q$ and $M \sim Qm$ up to $Q \lesssim (M_{\text{Pl}}/m)^2$. Above this value the system becomes relativistic (so that $\wp_{\text{quantum}} \sim Q/R^4$) and around the same value of $Q$ gravity also becomes strong enough to form black holes with mass $M \sim M_{\text{Pl}}^2/m$, which are thus lighter than in the fermionic case. The gravitational Bose balls would lie on a new DM line $R \sim (M_{\text{Pl}}/m)^2/M \propto 1/M$ in the 'catalogue' in fig. 3.1 with a downward slope that is parallel to the 'particles/waves' boundary, but well above this quantum boundary.

Instead of the gravitational force, the Fermi and Bose balls can be held together by stronger, model-dependent attractive forces. An attractive force among fermions, for instance, arises from tree level exchanges of a spin 0 scalar $s$ that has a Yukawa interaction with the fermion. A scalar with attractive self-interactions can form the so called $Q$-balls, which we discuss in more detail in section 10.5.3.

## 10.5.2 Cosmological formation of macroscopic dark bodies

Having discussed why it is possible that the macroscopic objects made out of dark matter particles could exist, we move on to the next issue: can such objects form during the cosmological evolution?

The answer to this question is yes, at least in the weakly coupled models where a DM particle (either a scalar, fermion or vector) acquires a mass equal to $m = gw$, where $w = \langle s \rangle$ is the vacuum expectation value of an extra scalar, $s$, and $g$ is the coupling between DM and $s$ [104]. We assume that initially the universe is in a false vacuum with $s = 0$, and that it contains relic DM particles with number density $n$ (this could possibly be due to a number asymmetry). Next, let us assume that $s$ during a first order phase transition at temperature $T \ll m$ acquires a nonzero vacuum expectation value $s$, and let us denote the energy density difference between the two vacua as $\Delta V$. During the phase transition, the bubbles of true vacuum with $w \neq 0$ appear, expand, and finally merge. Since inside the bubbles of true vacuum DM particles would have had a large mass $m$, well above the typical thermal kinetic energy of massless DM particles in the false vacuum, energy conservation prevents DM particles from entering the bubbles. The expansion of bubbles thus compresses DM into small remaining pockets of false vacuum, as illustrated in fig. 10.2. Each pocket contains a number $Q$ of DM particles, where the statistical distribution of $Q$

values can be computed in any concrete microscopic DM model. A variant of this mechanism that leads to microscopic DM candidates has been mentioned in section 4.5.2.

Of particular interest to us are the models in which *i)* the bubble walls move slowly, so that the pockets can dissipate thermal energy, and thus only a small fraction of DM particles escapes via thermal fluctuations, and *ii)* the annihilation rate of DM and $\overline{\text{DM}}$ particles inside the pockets is negligible (for example, because there is an asymmetry of DM vs $\overline{\text{DM}}$ abundances). The pockets of trapped DM particles then lead to formation of macroscopic objects. Compared to the discussion in the previous section, the energy of the system now has additional terms,

$$U = U_{\text{quantum}} + U_{\text{gravity}} + U_{\text{volume}} + U_{\text{surface}}, \tag{10.47}$$

where

$$U_{\text{volume}} = \Delta V \, \frac{4\pi R^3}{3}, \qquad U_{\text{surface}} = \sigma \, 4\pi R^2, \tag{10.48}$$

assuming for simplicity a spherical pocket, while $U_{\text{quantum}}$ and $U_{\text{gravity}}$ are given in eq. (10.45) above (in eq. (10.46) for the bosonic case). The volume term, $U_{\text{volume}}$, is proportional to the potential energy difference $\Delta V$ among the two vacua. Since it scales as $R^3$, it is typically larger than the surface term due to wall energy density $\sigma$, which only scales as $R^2$, and can thus be neglected for large $Q$. If the gravitational potential energy is also negligible, then $U \approx U_{\text{quantum}} + U_{\text{volume}}$. Since $U_{\text{volume}}$ increases with $R$, while $U_{\text{quantum}}$ decreases with $R$ (for fixed $Q$), there exists an equilibrium value of $R$. Its precise value depends on whether DM particles are fermions or bosons due to the different forms of $U_{\text{quantum}}$ in the two cases. However, in both cases the energy density is constant, $\rho = U/V \sim \Delta V$. That is, in this regime the DM particles trapped inside pockets of false vacua form objects of constant density, just like ordinary matter does, but for a completely different reason — in this case it is due to a constant energy density difference between false and true vacua, while in solids made out of ordinary matter it is due to almost uniform packing of massive building blocks — the atoms. Having assumed that the DM mass satisfies $m(0) \ll m(s)$, such macroscopic dark bodies have less energy than the free DM quanta would have had, if pockets were to be replaced by the true vacuum, and the dark bodies are thereby stable. DM particles inside the dark bodies are relativistic if $m(0) \ll 1/R$. On the other hand, if $m(0)$ is big enough so that $U_{\text{gravity}}$ is relevant for the stability calculation, the macroscopic dark objects can also gravitationally collapse and form either a black hole or a '*dark dwarf*'.[6]

While the above models may seem artificial, engineered such that one ends up with the remnant macroscopic dark bodies, the end result may in fact be more generic than one would have naively guessed. For instance, a very similar formation history of dark macroscopic objects may occur for gauge theories with 'dark quarks'. Assuming, for simplicity, a $\text{SU}(N_{\text{DC}})$ gauge dynamics with fermionic dark quarks, the non-perturbative interactions result in a first order confinement phase transition, if the number $N_F$ of fermionic quark flavours lighter than the confinement scale $\Lambda_d$ is either $3 \leq N_F \lesssim 3N_{\text{DC}}$ or $N_F = 0$ [104]. In the first case, all the scales relevant for the formation of dark objects are comparable: $\Delta V \sim \Lambda_{\text{DC}}^4$ is comparable to the latent heat, the wall pressure is $\sigma \sim \Lambda_{\text{DC}}^3$ and the dark hadron masses are of order $\Lambda_{\text{DC}}$. Pockets of false vacuum form with the initial percolation radius smaller than the Hubble length, $R_{\text{perc}} \gtrsim M_{\text{Pl}}^{2/3}/\Lambda_{\text{DC}}^{5/3}$. This non-trivial estimate is discussed, e.g., by Witten (1987) in [103] for the case of QCD; see also [104]. The pockets contain $Q \sim (nR_{\text{perc}})^3$ dark quarks, where $n$ is the thermal number density of quarks at the temperature of the phase transition. Individual dark quarks cannot enter the expanding confined phase (they can only leave the false vacuum pocket, if they form dark baryons in the process), and thus get compressed inside pockets until this reach equilibrium radius $R \ll R_{\text{perc}}$, where $R$ minimizes eq. (10.47). Only a fraction of dark quarks manage to exit the pockets, since a collection of free dark quarks in a false vacuum is lighter than a dark hadron with the same number of constituent dark

---

[6] With the term dark dwarf we denote an object that is physically analogous to a white dwarf (normal matter kept together by gravity and quantum pressure), but made out of dark matter.

quarks. In the same way as the mass of the proton is primarily set by $\Lambda_{\text{QCD}}$ rather than by the masses of $u$ and $d$ quarks, the mass of the dark hadron is primarily set by $\Lambda_{\text{DC}}$. Macroscopic balls of dark quarks are stable, if they are lighter than the corresponding dark baryons; in a given theory this depends on the fundamental parameters (such as the dark quark masses) and on the more uncertain non-perturbative factors (such as $\Delta V$). In that case we have, approximately,

$$Q \sim 10^{23} \left( \frac{\text{GeV}}{\Lambda_{\text{DC}}} \right)^3, \qquad M \sim \Lambda_{\text{DC}} Q \sim \text{gram} \left( \frac{\text{GeV}}{\Lambda_{\text{DC}}} \right)^2, \qquad R \sim \frac{Q^{1/3}}{\Lambda_{\text{DC}}} \sim \text{nm} \left( \frac{\text{GeV}}{\Lambda_{\text{DC}}} \right)^2, \qquad (10.49)$$

for dark quarks that are fermions, while if dark colored particles are bosons one has a somewhat smaller objects, $R \sim Q^{1/4}/\Lambda_{\text{DC}}$. Fig. 3.4 shows the bounds in the $(M, R)$ plane assuming that dark objects interactions with ordinary matter have cross sections of geometric size. Decoupled dark sectors will have smaller cross sections, down to gravitationally small size.

A similar discussion applies in the case of no light dark quarks (the $N_f = 0$ case). However, in that case also an additional, qualitatively different, possibility opens up: dark quarks can be heavy enough for gravity to become relevant in the calculation of properties of macroscopic objects, triggering further gravitational collapse of Fermi balls, creating either dark dwarfs or black holes (for large enough $m$).

### 10.5.3   Macroscopic objects held together by new forces: $Q$-balls

In the previous sections we considered macroscopic objects held together by universal forces: gravity, vacuum energy, quantum pressure. Instead of gravity, a new attractive force could be the reason the macroscopic bodies form. We bypass lengthy model-dependent discussions and focus on a rather simple case, the $Q$-balls made out of self-interacting scalars whose interactions are described by a fundamendal QFT [932]. The $Q$-balls themselves are also most easily descried in the language of field theory, where now the amplitude of the scalar field is related to the occupation number, and classical physics emerges when this is large (the same as the amplitude of the electromagnetic field is related to the number of photons). We will thus use the QFT language, at the risk of making $Q$-balls appear to be something deeper than a large number of particles held together by an attractive force. In this language $Q$-balls are described as particular configurations of the scalar field $\phi$. In contrast to some other stable field configurations, such as the dark monopoles discussed in section 9.6, topology plays no role in the analysis of $Q$-balls and their stability. The $Q$-balls are simply a bound state of many self-interacting scalar particles, which is stable because of their binding energy: the $Q$-ball has a smaller total energy than its constituents would have had, had they been free.

The simplest example of a theory in which $Q$-balls can form is a complex scalar field $\phi$ described by a Lagrangian invariant under a global U(1) symmetry, $\phi \to e^{i\delta}\phi$,

$$\mathscr{L} = |\partial_\mu \phi|^2 - V(\phi), \qquad V = m^2 |\phi|^2 + \lambda |\phi|^4 + \cdots. \qquad (10.50)$$

Stable $Q$-balls exist if the potential is such that $2V(\phi)/|\phi|^2$ has a local minimum at some field value $\phi_0$, which satisfies

$$\omega_0^2 \equiv 2V(\phi_0)/\phi_0^2 < m^2, \qquad (10.51)$$

where $m$ is the mass of a free $\phi$ particle. Above, we have also taken $\phi_0$ to be real, which can always be done without any loss of generality due to the U(1) symmetry.

Spherical field configurations of the form $\phi = \phi_0 f(r) e^{i\omega t}/\sqrt{2}$ have a global U(1) charge $Q$ and energy $E$ given by

$$Q = \int dV \; i(\phi^* \dot{\phi} - \phi \dot{\phi}^*) = \omega \phi_0^2 \int dr \; 4\pi r^2 f^2, \qquad E = \int dr \; 4\pi r^2 \left[ \frac{\phi_0^2}{2} (f'^2 + \omega^2 f^2) + V \right]. \qquad (10.52)$$

The profile $f(r)$ is determined by solving the classical equation of motion for field $\phi$, $(r^2 f')'/r^2 = \partial V/\partial f/\phi_0^2 - \omega^2 f$. One can show that $dE = \omega \, dQ$: $\omega$ encodes by how much the energy of the $Q$-ball varies when charge is varied. Generic configurations with a relatively small $Q$ have a smooth profile $f(r)$. The problem simplifies when $Q$ is large: the solution tends to $f = 1$ inside a sphere of radius $R$ and $f = 0$ outside (for details, see Coleman (1985) in [932]). In this limiting case the value of $R$ can then be found by minimising $E$, keeping $Q$ fixed, where now

$$Q = \omega \phi_0^2 \mathcal{V}, \qquad E = \left[ \frac{\omega^2 \phi_0^2}{2} + V(\phi_0) \right] \mathcal{V}, \qquad \mathcal{V} = \frac{4\pi}{3} R^3, \tag{10.53}$$

and we have already anticipated in the notation that the stable configuration will have constant $\phi$ inside the sphere equal to $\phi = \phi_0$, as we now show. That is, writing $\omega = Q/(\phi_0^2 \mathcal{V})$ gives $E = Q^2/(2\phi_0^2 \mathcal{V}) + V(\phi_0)\mathcal{V}$, which, when viewed as a function of $\mathcal{V}$, is minimised for $\mathcal{V} = Q/\sqrt{2\phi_0^2 V(\phi_0)}$, giving $E = Q\sqrt{2V(\phi_0)}/\phi_0$. The energy of the configuration is therefore minimal wherever $2V(\phi)/|\phi|^2$ has a minimum, i.e., at $\phi_0$ (at which point also $\omega = \omega_0$). Importantly, the value of $\phi_0$ is independent of $Q$, while the energy $E = Q\omega_0$ is directly proportional to the global $U(1)$ charge of the $Q$-ball. The $Q$-ball therefore behaves in a similar way as the ordinary matter, where the mass of a block of ordinary material is given by the number of constituents (or, equivalently, by the total baryon number). In particular, if one were to emit a single $\phi$ particle from a $Q$-ball, its energy would decrease by $\omega_0$, while the free particle at rest at infinity would add energy of its rest mass, $m$, so that for $m > \omega_0$ the $Q$-ball is stable.

The minimality condition in eq. (10.51) can be satisfied, for example, if $V$ has a local minimum away from the origin and $V(\phi_0) > 0$. So the theory has two minima, a local one at $\phi_0 \neq 0$, and a global one at $\phi_0 = 0$. This situation is close to (but not yet at) a first order phase transition, and is an example of an attractive self-interaction, with eq. (10.51) demanding that it is strong enough. In the limit where the two minima become degenerate, the $Q$-ball volume $\mathcal{V}$ diverges, meaning that the system goes through a phase transition. The $Q$-ball is thus a precursor of the new true vacuum, which after the phase transition encompasses the whole volume.

In a class of realistic models, the global $U(1)$ can be a baryon or a lepton number, in which case the existence of $Q$-balls becomes phenomenologically important for the details of baryogenesis. The scalar $\phi$ could be one of the scalars present in supersymmetric models (or, for instance, a linear combination of scalars that corresponds to a flat-direction in the super-potential); then a non-renormalizable term $\sim -|\phi|^6$ is needed to satisfy eq. (10.51) [932]. If $U(1)$ is gauged, the Coulomb potential energy prevents the existence of $Q$-balls carrying a macroscopically large charge $Q$. A nontrivial extension of the above discussion is, if the global symmetry is non-Abelian, which can then give rise to more complicated $Q$-balls.

There are several ways in which $Q$-balls can form during the cosmology evolution. If the scalar $\phi$ acquires a large enough vacuum expectation value beyond the potential barrier, the field can fragment during later stages of the cosmological evolution, resulting in the formation of $Q$-balls. Alternatively, first-order phase transitions can in in some models pack free particles into $Q$-balls, quite similar to the mechanism discussed in section 10.5.2 [104].

## 10.5.4   Axion stars

Another example of a macroscopic object whose constituents are possible DM candidates are *axion stars* (also known as *boson stars* or *Bose stars*) [933], which are made out of real scalars.[7] Perhaps the most celebrated example of such a light scalar is the QCD axion, reviewed in section 10.4, while another is the relaxion, discussed in section 10.1.6. More generally, axion stars can be formed by axion-like particles (ALPs, see section 3.4), whose interactions with the SM fields are of the same type as those of the QCD

---

[7]For the related case of *dark photon stars* (also known as *Proca stars*), see [934]. See also Liebling and Panzula (2012) in [934] for a comprehensive review of the different bosonic star possibilities.

axion but whose mass is treated as a free parameter. Axion stars are bound state configurations of ALPs, which are stabilised against the collapse by a coherent field gradient. This makes them distinct from the incoherently oscillating virialized systems such as the axion mini-clusters (section 10.4.4). Since there is no conserved particle number associated with real scalars, axion stars are metastable configurations and eventually decay.[8]

There are two distinct branches of possible axion stars, *i) dilute axion stars*, in the formation of which the gravitational potential energy plays an important consideration, and *ii) dense axion stars*, for which gravity is not important. **Dilute axion stars** have macroscopically large masses that depend on the number of bound ALPs, $Q$. They cannot grow arbitrarily large, though, since they become structurally unstable above a certain point. In this structural instability both gravity and the *attractive* force from the tiny ALP quartic negative self-coupling $V = m_a^2 a^2/2 - (g_4^2/4!)(m_a/f_a)^2 a^4 + \cdots$ plays a role [933]. The cosine-like QCD axion potential corresponds to $g_4 \sim 0.6$ [857]. Due to very large phase space density of axions, the small interactions can overcome the quantum pressure of eq. (10.46), triggering the collapse of the axion star. The energy of an axion star with radius $R$, consisting of $Q$ non-relativistic quanta, can be approximated as

$$U = U_{\text{quantum}} + U_{\text{grav}} + U_{\text{quartic}} \sim \frac{Q}{m_a R^2} - \frac{Q^2 m_a^2}{R M_{\text{Pl}}^2} - \frac{g_4^2 Q^2}{f_a^2 R^3}. \tag{10.54}$$

The first two terms were discussed in eq.s (10.46) and (10.45), respectively. The last term estimates the potential energy due to self interactions, $U_{\text{quartic}} \sim R^3 V_{\text{quartic}}$, in which the amplitude of the axion field $a$ was expressed in terms of the number of quanta $Q$ using the relation $Q m_a/R^3 \sim m_a^2 a^2$. Extremising $U$ with respect to $R$ shows that the stable gravitational Bose ball with negligible $U_{\text{quartic}}$, discussed below eq. (10.46), gets destabilised if $U_{\text{quartic}} \gtrsim U_{\text{grav}}$. This implies that axion stars heavier than $M_{\text{cr}} \approx 10 f_a M_{\text{Pl}}/m_a g_4$ are unstable. The order unity numerical factor comes from numerical studies [933]. For the QCD axion with $m_a \approx 10^{-5}$ eV this critical mass is $M_{\text{cr}} \approx 10^{-6} M_\oplus$, and the corresponding radius is $\sim 10^3$ km. The radius of an axion star scales as the inverse of the axion star mass, so that lighter dilute axion stars are spatially larger.

Due to their macroscopic size, the axion stars can have very long lifetimes. The microscopic process that leads to the slow decay of an axion star is the $3a \to a$ process, where the final state ALP has enough energy to escape. The lifetime of a dilute axion star scales roughly as $\tau \propto \exp(1/\Delta_a)$, where $\Delta_a = \sqrt{1 - E_a^2/m_a^2}$, with $E_a$ the typical total energy of a single bound ALP (slightly smaller than the rest mass of an ALP, $m_a$), and can thus be stable on cosmological time scales. For **dense axion stars** the binding is mainly due to the scalar field potential, with gravity unimportant. Typically, they also have short lifetimes, well below a second, and thus are not objects of interest for cosmology or astrophysics.

The end results of the axion star collapse depends on the details of the axion couplings. If the coupling to photons is large (about two orders of magnitude larger than in typical QCD axion models) a parametric resonant conversion of axions to photons is reached during growth of axion stars via Bose-Einstein condensation and thus all condensing axions are converted into photons with radio-frequency $m_a/2$, and an axion star never forms. For slightly smaller couplings the parametric resonance is never reached, an axion star forms, but during its subsequent collapse the conversion to photons is very efficient and results in a short but powerful burst of radio emissions.

Let us also recall that very light ALPs, with masses in the fuzzy DM regime, form gravitationally bound solitonic cores at the center of galactic halos, see section 6.14.

---

[8]This is distinct from $Q$-balls, discussed in the previous section, which are made out of complex scalars, and for which there is a conserved quantum number, so that the number of particles, $Q$, is conserved. Due to this conserved charge $Q$-balls cannot decay.

# 10.6 Origin of Dark Matter stability

DM needs to be stable on cosmological time scales. On the one hand, DM may be absolutely stable, usually due to some symmetry that forbids DM decays. Alternatively, DM may be unstable, but with a decay time that is simply much longer than the age of the Universe. An example of the latter possibility is a sterile neutrino DM, which is long-lived thanks to the combination of a small, keV-scale, mass, and the smallness of the mixing angle with the SM neutrinos, that induces the decay, see section 9.2.2. Another example are primordial black holes, which are long lived because their Hawking radiation is suppressed at large, asteroid-scale, masses. Apart from these special cases, theories of DM need mechanisms that highly suppress DM decays. Broadly speaking these fall into one of the following categories: an ad hoc symmetry (section 10.6.1), explicit symmetry present in the theory for other reasons (section 10.6.2), or accidental stability (section 10.6.3). For a dedicated discussion, see [935].

## 10.6.1 Ad hoc symmetries

The theoretically least appealing possibility is that the DM stability is put in by hand, i.e., by imposing a symmetry that is invoked just to prevent DM decays. The most commonly used symmetries are:

▶ **Discrete symmetries.** The most widely used possibility is the minimal one: a $\mathbb{Z}_2$ symmetry under which DM is odd, while any lighter particles are even, thus preventing DM to decay. Some DM models, though, rely on $\mathbb{Z}_3$ or higher $\mathbb{Z}_N$ symmetries for DM stability.

▶ **Global symmetries.** The simplest and also the most commonly used possibility is a global dark U(1) under which DM particles carry a nonzero charge. It implies a conservation of the associated DM number, and also allows for cosmological evolutions that lead to a non-zero DM asymmetry.

Gravity is expected to break global symmetries, both continuous and discrete. However the numerical size of the effect is unknown, and could be negligible for any practical purpose (in the same way as the baryon number in the SM is not exactly conserved, and is broken by sphaleron field configurations that are completely negligible at low temperatures).

## 10.6.2 Symmetries motivated by other reasons

The symmetry that guarantees the stability of DM could be a consequence of a bigger symmetry structure within a full theory, which apart from DM particle and the SM contains also other states.

▶ **Supersymmetric** models such as the MSSM assume a $\mathbb{Z}_2$ symmetry, *R*-parity, under which all the supersymmetric particles are odd and all the SM particles are even. The *R*-parity was introduced in order to avoid phenomenological problems: to ensure sufficient proton stability and to weaken bounds on new flavour-violating effects, see section 10.1.2 for further details. A side implication is that the lightest supersymmetric particle is stable and can thus be a DM candidate.

▶ Some of the possible dark gauge groups predict extended field configurations that are stable for **topological reasons**, and could thus be DM candidates. For instance, dark magnetic monopoles are stable for topological reasons in theories where a gauge group $\mathcal{G}$ is spontaneously broken to a sub-group $\mathcal{H}$, where $\mathcal{H}$ contains U(1) factors (section 9.6).

▶ *If* **something different** than the usual fermions and bosons can make sense within QFT, this something might only couple in pairs to the rest of the SM, and thereby be a stable DM candidate. Possible speculative ideas along these lines are:

▷ DM as a para-fermion (a tentative particle with intermediate quantum statistics) [936].

▷ DM as a dimension-1 'Elko' fermion, which is a non-local representation of the Lorentz subgroup spanned by the $J_z$, $K_z$, $K_x + J_y$, $K_y - J_x$ generators. The fact that it contains all $K_{x,y,z}$ boosts avoids the strongest bounds on Lorentz violation [937].

▷ In 1971 Dirac proposed a system of relativistic particles with mass $M$ and positive frequency only, described by an infinite-component wave equation [938]. This system of particles only (without anti-particles) received little interest because it cannot be consistently coupled to gauge fields, and also other SM fields. This property becomes favourable for a DM candidate [938].

▷ Gauge-invariant QFT are usually restricted to quantum states on which expectation values satisfy the classical equations of motion, such as the 1st Maxwell equation (in electromagnetism) and/or the 1st Friedmann equation (in general relativity). More general quantum states violate such classical equations, by containing electric or gravitational fields as those generated by extra stable charge and/or mass, despite that such sources are not present. This phenomenon can be identified as DM, up to the problem that cosmological inflation would dilute it [939].

▶ **Symmetry breaking** can leave a residual symmetry that protects DM stability.

▷ A simple possibility, which requires at least two matter fields extending the SM, is that DM carries a charge $+1$ under a dark $U(1)_d$ gauge group, which is spontaneously broken by a vacuum expectation value of a scalar with dark charge $+2$. This then leaves a $\mathbb{Z}_2$ subgroup of $U(1)_d$ unbroken. This kind of structure can arise from bigger groups that admit spinorial representations.

▷ The SM motivated $U(1)_{B-L}$ group can lead to DM candidates, assuming suitable fractional charges [940].

▷ The residual symmetry protecting DM stability can also occur in extensions of the SM by a single field. Such theories with a minimal field content can even be systematically classified. That is, a dark gauge group (SU($N$), SO($N$), Sp($N$) or exceptional groups) broken spontaneously to a sub-group $\mathcal{H}$ by a single scalar $S$ in a given representation sometimes results in acceptable DM candidates [941]. These theories have only a few free parameters, and predict a non-trivial cosmology and phenomenology. For instance, $\mathcal{H}$ can confine at energies below the spontaneous symmetry breaking scale, which can result in a composite DM. In some cases such DM candidates are stable at the renormalizable level, and the symmetries ensuring its stability are of accidental nature. This is discussed in section 10.6.3, together with the non-trivial mathematics of Lie groups.

## 10.6.3   Accidentally stable Dark Matter

An interesting possibility is that DM stability is due to an accidental symmetry, analogous to how the accidental baryon number explains the proton's quasi-stability in the SM (baryon number conservation in the renormalizable SM Lagrangian arises simply as a consequence of the assumed field content and gauge symmetries). A variety of accidental symmetries can similarly lead to a stable DM candidate:

▶ DM might be a **fermionic 5-plet** under weak SU(2)$_L$, as discussed in section 9.3.4.

▶ DM might be stable due to an accidental symmetry in a dark sector. For example, DM might be analogous to the proton: a composite particle under a new confining gauge group, stable thanks to an accidental U(1) **dark baryon number**. Another related possibility is that DM is a dark pion stable thanks to a dark flavour symmetry or due to dark $G$-parity, as discussed in section 9.5.

▶ DM stability could also be due to more profound versions of accidental symmetries of a group theoretical nature in the dark sector, for instance, due to automorphisms that some of the gauge groups admit. The simplest example of an automorphism is **charge conjugation**, for which the simplest realization is DM that is a dark U(1) gauge boson $A_\mu$, which obtains its mass after a scalar $S$ charged under the dark U(1) acquires a vacuum expectation value. Despite the symmetry breaking, the massive dark gauge boson remains stable, if the theory is invariant under charge conjugation in the dark sector,

$$S \to S^*, \qquad A_\mu \to -A_\mu. \tag{10.55}$$

In this simple model, charge conjugation is, however, not an accidental symmetry, because it can be broken by the kinetic mixing between hypercharge and dark U(1) [941]. Therefore an abelian vector $A_\mu$ is a viable DM candidate only if this mixing is sufficiently suppressed.

▶ A less minimal example is that DM particles are the three gauge bosons of an SU(2) dark gauge group that is completely spontaneously broken by a scalar $S$ in the fundamental representation 2 of SU(2) that obtains a non-zero vacuum expectation value. The three SU(2) vectors are stable DM candidates [941] because the theory is accidentally invariant under the *charge conjugation in the dark sector,*

$$S \to S^*, \qquad A_\mu^a T^a \to -(A_\mu^a T^a)^*, \tag{10.56}$$

where $T^a$, $a = \{1, 2, 3\}$, are the gauge group generators in the fundamental representation. The CP-odd stable vectors are associated with the real $T^a$ generators of the fundamental representation, namely the $T^{1,3}$ generators, in the usual notation. This theory also contains a larger accidental custodial global symmetry, such that the three vectors acquire a common mass $M$. The dark topological term, $A_{\mu\nu}^a \tilde{A}^{a\mu\nu}$, gives exponentially small CP-violating contributions for small values of gauge coupling.

More generally, complex conjugation is a non-trivial $\mathbb{Z}_2$ outer automorphism of the SU($N$), SO($2N$) and $E_6$ groups with symmetric Dynkin diagrams. So the accidental DM stability of the type discussed above can be realized in many different contexts.

▶ Some DM theories with a dark gauge group can be accidentally invariant under a $\mathbb{Z}_2$ discrete symmetry, which acts on components of dark multiplets rather than on full multiplets. An example is a $\mathbb{Z}_2$ under which only the $i = 1$ component of a field in the fundamental representation of the symmetry group is odd, while the remaining components are $\mathbb{Z}_2$ even. For other representations, the transformation rules can be built by considering products of fundamentals. For instance, dark vectors can be written as a matrix $(A^a T^a)_i{}^j$, and the $i = j = 1$ indices are taken to be odd. This symmetry is called **group parity** because it is analogous to space-time parity, except that it acts in group space. In a theory with an unbroken group parity, the lightest parity-odd state is stable. For instance, group parity can ensure the stability, for any $N$, of the SO($N$)$_{\rm DC}$ baryons, which are states built by contracting dark fermion fields $\Psi_i$ with the anti-symmetric tensor.

▶ Special groups with symmetric Dynkin diagrams contain extra special symmetries that can also ensure DM stability: SO(8) contains a **triality**; the exceptional group $\mathcal{G}_2$ an inner automorphism (see, e.g., Buttazzo et al. in [941]), etc.

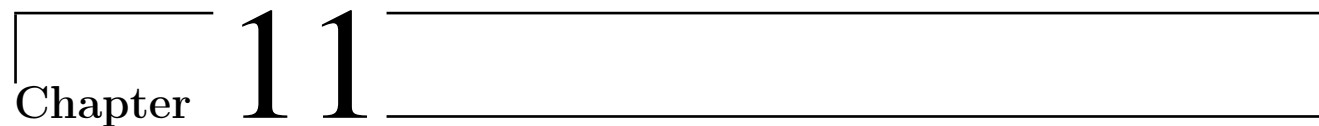

# Chapter 11

# Alternatives to Dark Matter

As reviewed in chapter 1, observations of a wide range of systems at vastly different scales provide compelling evidence for a discrepancy between the visible and inferred masses of these systems. Much of our discussion has centered on interpreting this discrepancy through the lens of an additional substance known as Dark Matter. In this final chapter we revisit these observations to discuss how and to what extent they can be understood through alternative interpretations that do not involve the introduction of extra matter.

The present puzzling mystery of the missing mass has some similarities to the one that the astronomy community was facing in the 1840s: the orbit of the planet Uranus, at the time the farthest known planet in the solar system, had been observed to violate the standard Newtonian laws of celestial mechanics. This led the french astronomer Urbain Le Verrier to postulate the existence of an extra planet whose gravitational effect would explain the anomalies in Uranus's behavior.[1] Such a planet, named Neptune, was indeed observed in 1846 by Johann Galle, almost exactly at the position indicated by Le Verrier on the basis of his computations [942]. During the same period, an additional anomaly was being discussed in the astronomy community: the planet Mercury was also displaying a puzzling feature not predicted by the Newtonian gravity — the precession of its perihelion. This led once again Le Verrier to interpret the observed deviation as the gravitational effect of yet another hypothetical planet, which was named Vulcain. Despite decades of searches and even some false discoveries, Vulcain was never found. Instead, this anomaly indicated new physics beyond the paradigm of the time: the anomaly was later understood by Albert Einstein as a correction to Newtonian gravity due to General Relativity [943].

The analogy with the topic of this review is evident: we are now in a similar situation since we only have indirect gravitational 'anomalies'. Are they due to some extra (dark) matter, as was the case for Neptune, or are they due to extra gravitational physics, as in the case of the non-existent Vulcain? So far, no convincing modification of gravity has been proposed which can reproduce all the anomalies described above. Hence, most of the community currently directs its efforts toward the first possibility, i.e., that there is additional matter. Nevertheless, DM has never been observed directly. Interesting alternative ideas have been proposed that can fit a sub-set of the gravitational anomalies. Below, we review these ideas.

## 11.1 Puzzling regularities in galactic rotation curves

Attempts to propose alternative ideas to DM started already early on, when galactic rotation curves were the main evidence for DM (a role played today by the formation of structures in cosmology). Galaxies

---

[1]Independently, John Adams, working at Cambridge in England, put forward the same idea, but his works were not published until later.

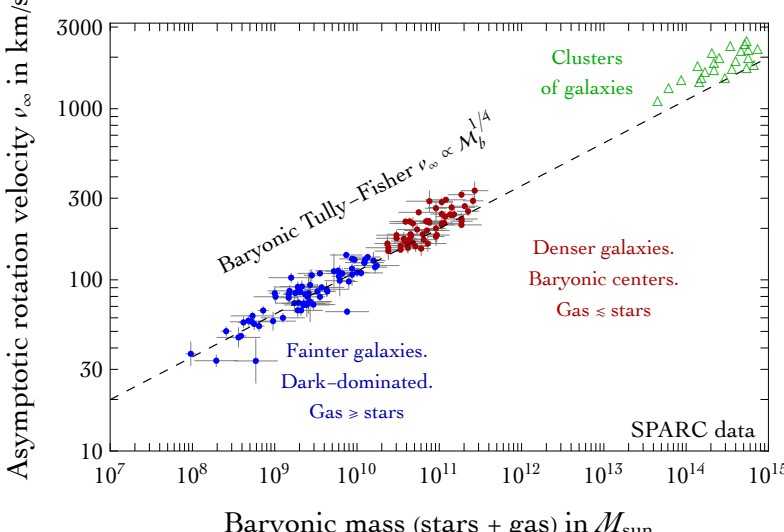

Figure 11.1:   ***Baryonic Tully-Fisher*** *correlation* $v_\infty \propto \mathcal{M}_b^{1/4}$ *among galactic baryonic mass* $\mathcal{M}_b$ *and rotation velocity* $v(r)$ *at asymptotically large radii* $r$. *Galaxies with larger (smaller) surface brightness are in red (blue). (SPARC galaxy data from* Lelli et al. (2019) *in [780]; cluster data from* McGaugh (2007) *in [945]).*

are now regarded as complicated but ordinary objects, and thereby as unlikely places where new fundamental physics might show up. More technically, attempts to explain galactic rotation curves through unusual modifications of gravitational dynamics generically fail when one adds new dependencies on size, density or curvature. This is because galaxies are not extreme objects in terms of these variables, hence modifications of General Relativity in galactic conditions typically lead to larger deviations elsewhere, in disagreement with observations. However, galaxies probe extremely small accelerations, opening the door for a modified dynamics in this regime.

## 11.1.1   Rotation velocities at asymptotically large radius

The first intriguing observation in this sense was that rotation curves asymptotically tend to a constant rotation velocity $v_\infty$. In the context of DM this regularity is understood to be due to the specific form of the density profile $\rho(r)$, that is produced by self-similar collapses, as discussed in section 2.1.1. Alternatively, this regularity was interpreted by M. Milgrom as a new physical law known as the Modified Newtonian dynamics (MOND) [944]. MOND postulates that below some critical acceleration $a_*$ the usual Newton law $F = ma$ changes to $F = ma^2/a_*$, where the quadratic dependence has been chosen such that the rotation curves become flat at accelerations below $a_*$. Using $a = v^2/r$ and spherical approximation, gives the scaling behaviours

$$\boxed{\frac{Gm\mathcal{M}_b(r)}{r^2} = F \simeq \begin{cases} ma & a \gg a_*, \\ ma^2/a_* & a \ll a_*, \end{cases}} \quad \Rightarrow \quad v(r) \simeq \begin{cases} (G\mathcal{M}_b(r)/r)^{1/2} & \text{Newton}, \\ (G\mathcal{M}_b(r)a_*)^{1/4} & \text{MOND}, \end{cases} \quad (11.1)$$

where $\mathcal{M}_b(r)$ is the *baryonic* mass inside a sphere of radius $r$. There is no DM in this interpretation. More formally, the flatness of rotation curves arises because the modified Newtonian dynamics $a^2 \propto 1/r^2$ is invariant under dilatations of space and time.

The above eq. (11.1) implied that the asymptotic rotation velocities at large $r$ should scale as a well-defined power of the galactic mass

$$v_\infty \propto \mathcal{M}_b^{1/4}. \tag{11.2}$$

The observational verification of this expression is known as the *baryonic Tully-Fisher relation*[2] [780]. The rotation curves have now been observed for about 200 galaxies, spanning about 4 orders of magnitude in $\mathcal{M}_b$: the data plotted in fig. 11.1 are compatible with the universal law given in eq. (11.2). The scatter around this prediction could well be due to experimental uncertainties, which are significant and can reach up to about a factor of 2.

In view of the significant diversity among galaxies, one could have expected a wider distribution. The heavier galaxies (plotted in red) have high surface brightness: their center contains a high density of baryonic stars, so that the rotation curves rise steeply up to the particular asymptotic constant value of $v$. The lighter galaxies (plotted in blue) have low surface brightness: their baryonic content is mostly in the form of a diffuse gas, and their rotation curves rise slowly. This can be seen from the rotation curves plotted in the center-right panel of fig. 1.2. Example of each of the two types of galaxies are also plotted in more detail in the bottom row in fig. 1.2 (see also fig. 2 in McGaugh & de Blok (1998) in [946]). The brighter galaxies have baryon-dominated centers, while the fainter galaxies are dominated by a dark component. Despite the differences, observations found that the fainter dwarf galaxies also follow the Tully-Fisher relation. This gave MOND a large boost in interest, and is arguably the most important successful prediction from MOND.

Independently from MOND, the observed baryonic Tully-Fisher relation means that the galactic rotation curves tend to flatten to a constant velocity at large radii, when the acceleration $a$ falls below the *common critical value*

$$\boxed{a_* \approx (1.2 \pm 0.1)\, 10^{-10}\, \mathrm{m/s^2}}\;, \tag{11.3}$$

numerically close, in natural units, to the present Hubble constant and to the inverse age of the Universe $H_0 \simeq 67.4\,\mathrm{km/(s \cdot Mpc)} \simeq 6.5\ 10^{-10}\,\mathrm{m/s^2} \sim 1/T_\mathrm{U}$. The above critical acceleration is common to almost all galaxies, even though galaxies have different sizes and masses. Equivalently, a common $a \approx G\mathcal{M}_b/r^2$ means that smaller galaxies have lower densities $\rho \propto \mathcal{M}_b^{1/2}$, which is also plotted as the correlation "mass $\propto$ size$^2$" in the catalogue in fig. 3.1.

## 11.1.2 Rotation curves at generic radii

The rotation curves contain more information than just that they are asymptotically flat. The observations indicate that *"for any feature in the luminosity profile there is a corresponding feature in the rotation curve"*, known as Renzo's rule (from Renzo Sancisi in [946]). This is causally understood in baryon-dominated regions, but it is also observed in regions dominated by DM, where it is unclear if correlations due to the formation history can explain this observation, given that one would expect DM to have a roughly spherical and smooth density.

Motivated by MOND considerations, the full rotation curves have been studied, in order to explore wether the radial gravitational accelerations $g_\mathrm{obs}(r)$, which have been measured in many systems, correlate with the acceleration $g_\mathrm{bar}(r)$ computed based on the observations of baryons alone [945]. The results appear to be consistent with the MOND expectations: the large spread between different rotation curves

---

[2]The earlier Tully-Fisher relation, $L \propto v_\infty^4$, is an approximate empirical correlation between the intrinsic total luminosity of the galaxy, $L$, and the rotational velocities, $v$, on the outskirts of spiral galaxies. The proportionality constant is found to be nearly universal, independent of the type and the size of the galaxy. The luminosity $L$ is a measure of the total *stellar* mass in the galaxy. Summing the stellar mass and the matter in the form of gas, gives instead the total *baryonic* mass $\mathcal{M}$ in the galaxy. The *baryonic* Tully-Fisher relation, $\mathcal{M}_b \propto v_\infty^4$, has a smaller observational scatter in the values of the proportionality constant for different galaxies than does the scatter in the Tully-Fisher relation $L \propto v_\infty^4$.

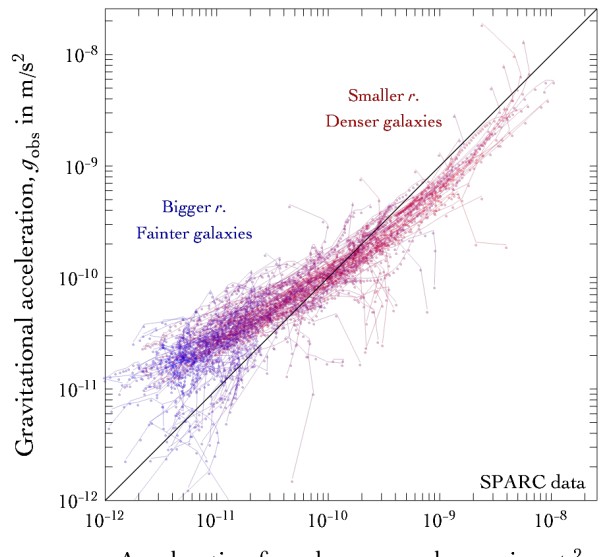
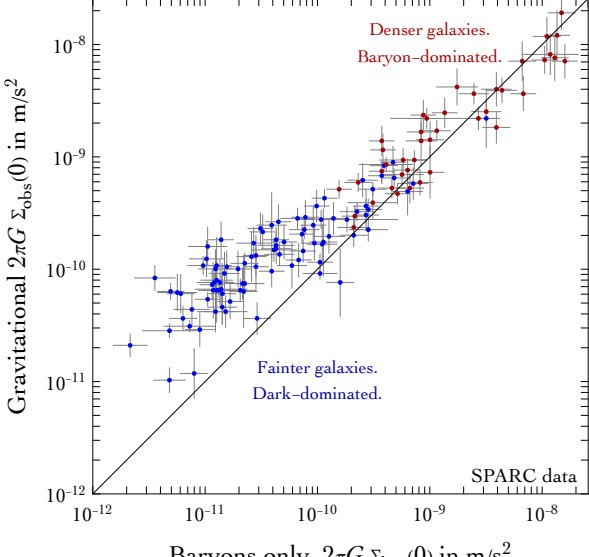

Figure 11.2:    **Left**: *correlation between the centripetal acceleration* $g_{\mathrm{obs}} = v^2/r$ *observed in galactic rotation curves and the gravitational acceleration* $g_{\mathrm{bar}}$ *computed from baryons only. Galaxies with larger (smaller) surface brightness are in red (blue). (SPARC galaxy data from [945]). Weak lensing techniques try extending data down to* $g_{\mathrm{bar}} \sim 10^{-15}\,\mathrm{m/s^2}$ *finding rough agreement with the MOND expectation* $g_{\mathrm{obs}} \sim \sqrt{g_{\mathrm{bar}} a_*}$, *see KiDS-1000 in [945].* **Right**: *correlation between the surface density* $\Sigma(0)$ *in galactic centers as measured from baryons only and as measured from gravity. (SPARC galaxy data from [947]).*

plotted in the middle-right panel in fig. 1.2 collapses (within the significant uncertainties) onto a common curve in fig. 11.2a, which also shows an indication of a possibly universal deviation from the $g_{\mathrm{obs}} = g_{\mathrm{bar}}$ equality, below the expected critical value of the acceleration.

The near equality $g_{\mathrm{obs}} \approx g_{\mathrm{bar}}$ that is observed at large accelerations is a result of standard physics and is predicted both by MOND and DM; the accelerations are large in the inner regions, which are dominated by baryonic matter and thus largely insensitive to DM. The question on the other hand is, whether the tight correlation between $g_{\mathrm{obs}}$ and $g_{\mathrm{bar}}$ at lower accelerations can arise in the DM context, namely, whether $\Lambda$CDM predicts similar enough formation histories for the various galaxies.

### 11.1.3   Rotation velocities at small radii

Baryonic matter, distributed on a thin galactic disk of surface density $\Sigma(r)$, generates a Newton potential $\phi(r, z)$ according to the Poisson equation. Rotation curves $v(r)$ on the galactic plane satisfy $v^2/r = -\partial\phi/\partial r|_{z=0}$. By writing the potential $\phi$ as a Fourier integral over the basis functions $\phi_k = J_0(kr)e^{-k|z|}$, one can show that the surface density at the galactic center, $\Sigma(0)$, can be written as a weighted integral over rotation curves (Toomre (1963) in [947])

$$2\pi G \Sigma(0) = \int_{-\infty}^{+\infty} \frac{v^2(r)}{r} d\ln r. \tag{11.4}$$

The above integral is dominated by the small $r$ region, so that the observed $\Sigma_{\mathrm{obs}}(0)$ is mostly determined by the measurements of the rotation curves in the smallest $r$ bins. This can then be compared with the measurements of the surface density close to the galactic center due to just the baryons, $\Sigma_{\mathrm{bar}}(0)$.

After correcting for the deviations from the thin disk approximation, one finds the results plotted in fig. 11.2b. Since these are effectively just a different way of combining the "generic $r$" results, discussed in section 11.1.2, it is not surprising that the two lead to very similar conclusions (compare the two panels in fig. 11.2). Galaxies with higher surface density again have baryon-dominated centers, so that $\Sigma_{\rm obs} \approx \Sigma_{\rm bar}$, while in galaxies with lower surface density one finds $\Sigma_{\rm obs} > \Sigma_{\rm bar}$. This indicates an extra dark component below a critical density $\Sigma_{\rm cr}$, corresponding to the critical acceleration $2\pi G \Sigma_{\rm cr}$, roughly compatible with eq. (11.3). Once again, the deviation from the predictions of the baryonic Newtonian gravity is compatible with a roughly-universal MONDian dynamics [947].

## 11.2 Can DM explain observed regularities in galaxies?

The observed regularities are intriguing and demand an explanation. Can they be understood in terms of DM, or do they signal a new universal phenomenon that is not due to DM?

### 11.2.1 Flat rotation curves from DM?

First, let us discuss the possible explanations within the DM framework. To understand better the problem, let us start from an over-simplified picture of a galaxy, taking baryonic matter of total mass $\mathcal{M}_b$ to be concentrated in the center, surrounded by an extended DM halo. For simplicity, let us assume that the DM halo at large enough radii has a density profile described by a simple power law $\rho(r) \sim k_{\rm DM}/r^s$. At small radii, where baryonic mass dominates, the rotation curves are given by $v = (G\mathcal{M}_b/r)^{1/2}$. At large radii the enclosed mass is dominated by the DM halo, and thus the rotation curves are given by $v_\infty = \sqrt{k_{\rm DM}G}\, r^{1-s/2}$.

Flat rotation curves, $v_\infty = \sqrt{k_{\rm DM}G}$, are then obtained for $s = 2$. The transition from baryon-dominated to DM-dominated gravitational pull happens at the critical radius $r_{\rm cr} \sim (\mathcal{M}_b/k_{\rm DM})^{1/(3-s)}$ which encloses equal amounts of mass in baryonic matter and in DM. At the critical radius the gravitational acceleration is $a_* \sim G\mathcal{M}_b/r_{\rm cr}^2$ which for $s = 2$ gives $a_* \sim Gk_{\rm DM}^2/\mathcal{M}_b$.

Can the clustering of DM form structures that have the correct density profiles $\rho(r)$ to reproduce flat rotational curves? In section 2.1.2 we obtained the slope $s = 2$ (see the discussion around eq. (2.8)) from the iso-thermal sphere approximation to DM clustering, while the more realistic DM density profiles discussed in section 2.2 and table 2.1 have scale-dependent slopes $s \sim 1-3$. That is, DM does tend to give "flattish" rotation curves, which agree well with data. A potential exception to this general agreement may be the measurements of gravitational potential around isolated galaxies using weak gravitational lensing [948], which, if confirmed, would imply flat circular velocity curves that remain flat up to large distances, even up to $r \sim$ Mpc and thus well beyond the expected virial radii of dark matter halos.

### 11.2.2 Baryonic Tully-Fisher relation from DM?

The amount of dark and baryon matter changes from galaxy to galaxy, while data indicates that $a_*$, which relates the two quantities via $a_* \sim Gk_{\rm DM}^2/\mathcal{M}_b$ (see above), does not. To agree with the observations the DM halo mass (controlled by $k_{\rm DM}$) and the baryonic mass $\mathcal{M}_b$ for each galaxy therefore need to be related according to $\mathcal{M}_b \propto k_{\rm DM}^2$. This is equivalent to the baryonic Tully-Fisher relation $v_\infty \propto \mathcal{M}_b^{1/4}$ [780], since the relation $\mathcal{M}_b \propto k_{\rm DM}^2$ implies that the flat rotation velocity, $v_\infty \sim \sqrt{k_{\rm DM}G}$, scales as $\mathcal{M}_b^{1/4}$.

The challenge for DM is whether this non-trivial relation between two seemingly unrelated quantities can be explained dynamically through the process of structure formation, reproducing the common behaviour observed across a wide range of galaxies. The answer to this question remains unclear [946]. In the minimal $\Lambda$CDM model the galaxy formation histories do have a common element: normal matter falls into potential wells that were already formed by cold DM. This does induce correlations between the

distributions of DM and normal matter, though very naively one might have guessed that it would have predicted a universal proportion between matter and DM. The Tully-Fisher relation instead demands relatively less baryons in smaller galaxies.

The approximate analytic description of the gravitational collapse of collision-less DM, described in section 2.1.2, predicts that the density of DM halos is controlled by the average cosmological DM density at the time the individual halo collapsed. Neglecting the effects of cosmological evolution, this would then give DM halos that are characterised by a common density $\rho$. Estimating the size $r$ of the galactic halo from $\mathcal{M} \sim \rho r^3$, and using it in the velocity curve relation $v^2/r = G\mathcal{M}(r)/r^2$ then gives the estimate $v_\infty \sim [\sqrt{G\rho}\mathcal{M}]^{1/3}$. Taking $\rho$ to be common to all galaxies, this would then imply a scaling $v_\infty \propto \mathcal{M}^{1/3}$, which clearly differs from the Tully-Fisher one, $v_\infty \propto \mathcal{M}_b^{1/4}$. Now, many crude approximations went into the above naive estimate. For one, the galaxies formed only relatively recently, at the redshift $z \sim 1$. Taking into account that the inhomogeneities on different scales grow to become non-linear at different times (see, e.g., section 2.2.5) results in additional scale dependence, which tends to improve the agreement with observations. Combining this dependence with the effects of baryon collapse, at least within a simple model, see, e.g., Kaplinghat & Turner (2001) in [946], then gives a possible explanation of how a seemingly universal value of $a_*$ might arise dynamically, at least for larger galaxies, which have their central regions dominated by baryons.

However, as argued by Milgrom (2001) in [946], the same $a_*$ is also seen in larger structures as well as in smaller galaxies, in which DM dominates over baryons and accelerations are below $a_*$. The regularities observed in galaxies seem to go well beyond the universality of DM halos. They appear to involve baryons in a key way, even though baryons constitute only a small fraction of the total apparent mass of the halo, and are often concentrated in a disk that is an order of magnitude smaller in size than the DM halo. Since the fraction of baryons in stars vs. gas seem to vary with galaxy size, this suggests that complex astrophysical phenomena are at play. For example, baryonic matter gets partially ejected by astrophysical effects such as the gas heating and supernova explosions, where the effectiveness of these processes depends on the escape velocity of the specific galaxy. This complexity renders the process of galaxy formation a challenging problem, and one that is not yet under good theoretical control. In the DM framework this complex astrophysics is presumably responsible for the observed regularities, with an important role played by the fact that smaller galaxies have a relatively lower baryon content (examples are shown in the bottom row in fig. 1.2).

Present numerical simulations and semi-analytic descriptions of galaxy formation within $\Lambda$CDM do not yet firmly differentiate between $v_\infty \propto \mathcal{M}_b^{1/4}$ and $v_\infty \propto \mathcal{M}_b^{1/3}$ scalings, and are also unable to definitively answer whether the simulated structures have low enough scatter in the seemingly universal laws to be compatible with the observations [946]. It may well be that both the flatness of the galactic rotation curves at large radii as well as the baryonic Tully-Fisher relation can be explained by a dynamical interplay between baryonic matter and DM during the process of galaxy formation. If this is indeed the case, then the observed regularities do not have any profound implications for fundamental physics. Instead, they would merely be a consequence of how normal matter and DM tend to cluster, with no deeper meaning. The opposite is also possible; the fact that regularities have been observed despite the complex astrophysical environments is considered by some to point to a new fundamental law, the MONDian dynamics.

The number of measured rotation curves will increase by at least an order of magnitude by the end of the 2020s, thanks to the ASKAP and the future SKA (Square Kilometre Array) telescopes. Time will tell, if future data will remain compatible with such universal laws.

## 11.3   Can MOND explain the observed irregularities?

While MOND can elegantly explain the flat rotation curves and the baryonic Tully-Fisher relation, it does face difficulties in several other systems, in which the clear-cut MOND predictions appear to be violated. Some of these difficulties are, roughly ordered in the increasing degree of seriousness, as follows.

i) On sub-galactic scales, some **globular clusters** of stars, such as Palomar 14 and NGC2419, appear not to follow the MOND predictions, even though they are located in the outermost regions of the Milky Way halo and therefore in low-acceleration conditions where MOND should apply [949]. The statistical significance of these observations is, however, under debate. For instance, it is possible that the globular clusters are on a highly eccentric orbit around the Milky Way, i.e., that they only 'recently' entered the low-acceleration regime, and thus have not yet reached the limit of MOND dynamics. Similarly, if the systematic uncertainties in the determination of the properties of the globular clusters were underestimated this could also ease the tension with MOND. To settle the issue in favour of DM one would need to observe galaxies with an inner region dominated by normal matter, an intermediate region in the MOND regime where rotation curves are instead Keplerian, before reaching the outer DM-dominated region with flat rotation curves.

ii) On super-galactic scales, clusters of galaxies are among the first systems where the missing mass problem has been identified (see section 1.2). Since the accelerations within clusters are comparable to $a_*$, clusters of galaxies are also good testbeds for MOND. Over the years, a number of analyses found that **MOND can only partially account for the missing mass in clusters** [949]. MOND predictions could in principle still agree with the data in the unlikely case that the observations have missed collision-less objects formed out of ordinary baryons (such as compact clouds) which would then have added up to the residual missing mass in the clusters.
The problem with MOND's predictions is exacerbated by the observations of the **bullet cluster**, as well as by observations of the other collisions among galaxy clusters (section 1.2.1). These show that whatever substance is responsible for 'dark matter', it is spatially separated from visible matter. It is thus not simply tied to the distribution of normal matter, contrary to the MOND postulate. (In order to explain the bullet cluster result within MOND, some authors initially suggested adding 2 eV neutrinos, a possibility that is now excluded [949].)

iii) Similarly to the bullet cluster, the **DM-deficient galaxies** discussed in section 1.1.3 would in principle constitute an example in which normal matter does not follow MOND predictions. MOND theories predict a modification of gravity at large enough distances such that $a \lesssim a_*$. However, the challenge is finding DM-deficient galaxies in this regime.

iv) On cosmological scales, the evidence for dark matter does not come merely from the galactic rotation curves, but most importantly from cosmological observations of the **CMB and structure formation** (as discussed at length in section 1.3), i.e., from several different length scales and accelerations [950].[3] Note however that, according to McGaugh et al. (2024) [781], observed galaxies formed faster than what Dark Matter predicts, in agreement with MOND.

From a more conceptual point of view, note that the phenomenological MOND postulate, eq. (11.1), is not a consistent theory. Such an empirical law, for instance, violates momentum conservation: the forces between two different masses are not equal and opposite, given that the lighter mass is subject to a bigger acceleration, and therefore to a bigger MOND effect. Over the years, several different internally

---

[3]Regarding other possible tests on large scales it is worth mentioning that some researchers believe the observed cosmological voids to be predicted to be much rarer within the standard cold DM cosmology, while their existence can instead be understood within MOND, supplemented, however, by some amount of DM in the form of 11 eV sterile neutrinos [949].

consistent theories have been proposed, which lead to MOND-ian dynamics in particular limits. We will review theories motivated by MOND in section 11.4.

These MOND-motivated theories can differ from the minimal phenomenological MOND postulate, predicting extra effects that can alleviate its difficulties, or lead to new ones. For example, it has been argued that the 'DM deficient' tension, mentioned in point iii) above, can be alleviated, at least in some systems, by the '*external field effect*' (EFE) acting on a small galaxy immersed in a massive host. Indeed, in theories of modified gravity, external sources can contribute more to the gravitational field than they do in Newtonian gravity. Chae et al. (2020) in [949] claimed a statistical detection of the EFE predicted by MOND theories. Furthermore, Kroupa et al. (2022) in [949] interpreted some asymmetric tidal tails of star clusters as EFE, and thus a challenge to Newtonian dynamics.

The small-acceleration MOND proposal can be tested directly by measuring **wide stellar binaries** away from external fields. Accelerations in these systems are so low that the MONDian theories predict the orbital velocities to receive an order unity enhancement compared to the Newtonian dynamics. The DM framework instead predicts that there will be no deviations from Netwonian physics, given that only little amounts of DM mass would be involved. Since the orbital periods of these systems are very long, the orbits are not fully reconstructed by observations. However, about 10000 such systems are partially observed by GAIA, allowing for statistical tests. The interpretation of data requires modeling of the undetected companions and of the uncertain eccentricity distributions of the wide binaries. The present situation is confusing; while some authors that analysed the GAIA data claim that wide binaries agree with the Newtonian dynamics, and thus exclude MONDian effects with up to $19\sigma$ significance (see Banik et al. (2023) in [951]), others instead claim a discovery of MONDian anomalies with a significance of up to $10\sigma$, and thus an exclusion of Newtonian dynamics (see Chae (2023) in [951]).

In the **solar system**, the solar gravitational acceleration is about $10^4 a_*$ around Pluto's orbit, and becomes smaller than $a_*$ at distance $r \approx 0.1\,\mathrm{lyr}$ from the Sun. Vokrouhlicky et al. (2024) [949] claim that simulations of comets and planetesimals around the Oort cloud at distances $10^{-3} - 1$ light years from the Sun prefer Newtonian dynamics over its MOND modification at $a \lesssim a_*$. Full MOND theories where this modification is screened by the EFE are not disfavoured. However, Desmond et al. (2024) in [949] claim that such MONDian theories predict a quadrupolar correction to the Sun gravitational potential, which was not observed in the precise Cassini measurements of Saturn's orbit.

## 11.4   Theories of Modified Newtonian dynamics

All the theories discussed so far, most notably in chapters 9 and 10, add to the Standard Model of particle physics some extra field that, as far as observables in cosmology and astrophysics are concerned, behaves as stable non-relativistic particles: Dark Matter. Let us now turn our attention to the theories in which modifications of gravity account for at least some of the observations that are usually interpreted in terms of DM [952]. As we will see, it is hard to bypass cold DM or something similar to it, and match all observations.

As discussed around eq. (11.1), galactic rotation curves motivate a specific modification of Newtonian gravity that kicks in on galactic scales — MOND [944]. That is, flat rotation curves compatible with observations are obtained, if at low accelerations $a \ll a_* \approx H_0$, the equation of motion for a test particle is modified from its form given by Newtonian gravity, $a = GM/r^2$, to a MONDian form, $a^2/a_* = GM/r^2$ (with $a_*$ given in eq. (11.3)).

This change can be accomplished either by modifying inertia or gravity. A modification of inertia of the right magnitude could, for instance, occur by having the existence of a cosmological horizon affect the inertia of bodies through a Hubble-scale Casimir-effect (see, e.g., Milgrom (1999) in [952]). In effect, one postulates $\sim H_0$ as the minimal acceleration, based on the observation that, at low accelerations, the horizon of the coordinates for a uniformly accelerating reference frame in flat spacetime (Rindler

coordinates) becomes comparable to the cosmological horizon (see, e.g., McCulloch (2010) in [952]). Developing the heuristic arguments for $a_* \sim H_0$ into a full theory has, however, proved to be quite challenging.

A related highly non-standard proposal is that gravity might be an emergent phenomenon, related to entanglement entropy ('**emergent gravity**' [953]), and that this could explain Newtonian gravity together with a MOND-like modification below a critical acceleration that is comparable to the current acceleration of the Universe. It is not yet clear whether this idea can reproduce relativistic gravity (tested in environments such as the solar system, dwarf galaxies and galaxy clusters [954]) and the observed cosmological CMB anisotropies. A different non-standard proposal assumes that gravity behaves classically up to stochastic corrections that might mimic a minimal accelerations similar to what postulated by MOND [955].

Modifications of gravity, more precisely the gravitational acceleration $g$, have been proposed in several tentative realizations using field theories. These introduce extra fields, very much like in DM theories. The first step in this direction was the construction of a non-relativistic field theory for MOND by Bekenstein and Milgrom (1984) in [952], who modified the Poisson equation for the non-relativistic Newton potential $\phi$ at small gravitational fields, $x \equiv (\nabla \phi)^2/a_*^2 \ll 1$, by assuming a non-relativistic Lagrangian density of the form

$$- \mathscr{L}_{\mathrm{MOND}}^{\mathrm{non\ rel}} = \frac{a_*^2 J(x)}{8\pi \tilde{G}} + \phi \rho, \qquad \text{implying} \qquad \nabla \cdot \left[ J'(x) \nabla \phi \right] = 4\pi \tilde{G} \rho, \qquad (11.5)$$

where $\rho$ is the baryonic mass density and $J(x)$ is some function. Newtonian gravity is obtained for the usual quadratic action, $J(x) = x$ and $\tilde{G} = G$. MOND is obtained if $J'(x) \simeq \sqrt{x}$ in the limit $x \ll 1$, corresponding to $J(x) \simeq \frac{2}{3} x^{3/2}$. In this limit a point mass $M$ sources $\phi \simeq \sqrt{GMa_0} \ln r$ and thereby an acceleration

$$\boldsymbol{a} = -\nabla \phi = -\sqrt{\tilde{G} M a_*} \, \boldsymbol{r}/r^2 \qquad \text{at} \qquad r \gg r_* \equiv \sqrt{\tilde{G} M/a_*}, \qquad (11.6)$$

resulting in the desired $v(r)$ dependence of galactic rotation curves, eq. (11.1). The above theory extends the MOND postulate beyond spherical or cylindrical symmetry, and respects momentum conservation, since its action is invariant under translations. However, it is neither Galilean nor Lorentz invariant: gravity is corrected in a way that is vaguely analogous to how a dielectric modifies equations of electromagnetism, so that the physics depends on the absolute acceleration with respect to the special 'rest' frame. This gives the '*external field effect*', already anticipated in section 11.3: forces within a system that is embedded in a bigger external system (such as a galaxy in a cluster of galaxies) also depend on the gravitational acceleration $\boldsymbol{g}_{\mathrm{ext}}$ of the latter. In contrast, Newtonian and Einstein gravity give rise to tidal forces only, related to spatial gradients of $\boldsymbol{g}_{\mathrm{ext}}$, and thus respect the strong equivalence principle.

Since theories of gravity are strongly constrained by general covariance, it is convenient to also consider another class of MOND theories: those that can avoid modifying the action for the Newtonian potential $\phi$. Keeping its usual action, this can be obtained by adding a scalar $\theta$ that kinetically mixes with $\phi$, giving the Lagrangian

$$- \mathscr{L}_{\mathrm{MOND}}^{\mathrm{non\ rel}} = \frac{1}{8\pi \tilde{G}} \left[ [\nabla(\phi - \theta)]^2 + a_*^2 J(y) \right] + \phi \rho, \qquad y \equiv \frac{|\nabla \theta|^2}{a_*^2}. \qquad (11.7)$$

There are various choices for function $J$, which smoothly interpolate between the Newtonian and MOND limits. To avoid conflicts with precision solar-system measurements (such as the precession of orbits), one needs a function $J$ that quickly reduces to Newtonian gravity in its regime of validity, such that MOND effects are strongly suppressed (see, e.g., Skordis (2009) in [952]).

### 11.4.1   MOND as tensor-vector-scalar gravity

The formulation of MOND in eq. (11.7) paves the way for the next step; to go beyond the small-velocity and weak-field limits, and find a MONDian extension of general relativity. The best known extension of this sort is the TeVeS construction by Bekenstein (2004) in [952]. In TeVeS, the Newtonian potential $\phi$ is, as in the usual general relativity, given by the $g_{00}$ component of the Einstein metric $g_{\mu\nu}$, $g_{00} = 1 + 2\phi$, while the scalar $\theta(\boldsymbol{x})$ becomes a shift-symmetric Lorentz scalar $\theta(x_\mu)$. In order to obtain from a covariant action the peculiar MONDian action with the unusual spatial gradients, an extra four vector $U_\mu$ with time-like unit norm, $U_\mu U^\mu = 1$, is introduced (thus the TeVeS acronym, which stands for the Tensor-Vector-Scalar gravitational field content). The main role of $U_\mu$ is to be able to induce the desired mixing of the scalar $\theta$ with the Newtonian gravitational potential $\phi$ (for the details see below). The field $U_\mu(x)$ has a non-vanishing vacuum expectation value $\langle U_\mu \rangle$ along the time direction, which then selects a special frame and breaks the strong equivalence principle.[4]

In this initial version of TeVeS, ordinary SM fields couple to a modified metric $\tilde{g}_{\mu\nu}$, which is a function of $g_{\mu\nu}$, $U_\mu$, and $\theta$. Consequently, these theories predicted that the electromagnetic and gravitational waves would have different speeds of propagation. This prediction was contradicted by the simultaneous detection of gravitational waves from a neutron star merger (GW170817) and its electromagnetic counterpart in the form of a gamma-ray burst. The close arrival times of the two signals limited the allowed difference in the speeds of photons and gravitons to be less than about one part in $10^{15}$, excluding TeVeS and similar theories of MOND [956].

Putative MOND-ian extensions also face challenges with cosmological measurements. The main challenge is how to mimic the cosmological imprints that are a result of a two-component fluid in the standard $\Lambda$CDM cosmology: the noninteracting DM fluid and the interacting plasma composed of normal matter and photons. As discussed in detail in section 1.3.1, the two components behave differently and thus also evolve differently. The fluctuations in DM evolve under the influence of gravity, while the fluctuations in the baryon-photon fluid oscillate as sound waves. After recombination, the baryons decouple from photons and fall into the growing gravitational potential wells due to DM. This process erases most of the signatures of the sound waves, and explains why there are $\mathcal{O}(1)$ oscillations seen in the CMB (section 1.3.3), and much smaller ones in the distribution of galaxies, $\mathcal{O}\big(\Omega_b^2/\Omega_{\mathrm{DM}}^2\big) \sim 0.04$. The MOND-ian theories need to achieve this suppression of acoustic fluctuations [950] without introducing a non-interacting fluid, such as a gas of cold DM particles. It has been suggested that a MOND cosmology that contains sterile neutrinos (which are too hot to be a cold DM) might agree with the CMB data, although structure formation might be problematic [950].

Skordis and Zlosnik proposed a relativistic theory of MOND that is claimed to solve the above problems (Skordis and Zlosnik (2019) and (2020) in [952]). The Lagrangian of the theory contains the SM fields coupled to $g_{\mu\nu}$, in the same way as in general relativity, and a modified gravitational sector

$$\mathscr{L}_{\mathrm{MOND}}^{\mathrm{TeVeS-SZ}} = \frac{1}{16\pi\tilde{G}} \left[ R - \lambda(U_\mu U^\mu + 1) - \frac{K_B}{2} F^{\mu\nu} F_{\mu\nu} + (2 - K_B)[2J^\mu(\nabla_\mu \theta) - Y] - F(Q, Y) \right] + \mathscr{L}_{\mathrm{SM}},$$
(11.8)

where $\lambda$ is a Lagrange multiplier that fixes $U^\mu$ to be a unit time-like vector, with $F_{\mu\nu} \equiv 2(\nabla_\mu U_\nu - \nabla_\nu U_\mu)$, $J^\mu \equiv U^\alpha \nabla_\alpha U^\mu$, while $F$ is some function of

$$Y \equiv (g^{\mu\nu} + U^\mu U^\nu)(\nabla_\mu \theta)(\nabla_\nu \theta) \qquad \text{and} \qquad Q \equiv U^\mu \nabla_\mu \theta.$$
(11.9)

To see how the theory works, let us specialize to the cases of dynamics on the galactic scale and to cosmology:

---

[4]In the $\Lambda$CDM framework the cosmological evolution instead selects a special frame, in which the Dark Matter and ordinary matter are at rest, on average.

▷ In the non-relativistic limit $Y$ reduces to $(\boldsymbol{\nabla}\theta)^2$, and $J^\mu(\nabla_\mu\theta)$ to $(\boldsymbol{\nabla}\phi)\cdot(\boldsymbol{\nabla}\theta)$. The theory then reduces to the desired form in eq. (11.7) plus an additional quadratic "mass term" for the Newton potential $\phi$. This undesired term is proportional to $\dot\theta$, and must be smaller than $\sim 1/\mathrm{Mpc} \sim 10^{-30}\,\mathrm{eV}$ in order to reproduce the galactic rotation curves *à la* MOND. The critical acceleration $a_*$ depends on the scalar vacuum expectation values, and $a_* \sim H_0$ is not a generic outcome.

▷ The cosmological time evolution of $\theta$ allows the scalar $\theta$ to play the role of DM in cosmology. Let us consider the homogenous limit where $\theta$ depends only on time. This corresponds to $Y = 0$ and $Q = \dot\theta$, so that the scalar Lagrangian reduces to $\mathscr{L} = -F(Q,0)/16\pi\tilde{G}$. Assuming a kinetic energy for $\theta$ minimal at $\dot\theta \neq 0$ gives a kinetic analogue of the Higgs phase, known as the 'ghost condensation' or the '$K$-essence'. The classical equation of motion, $\ddot\theta F'' + 3HF' = 0$, is solved by $F' \propto 1/a^3$. Let us assume that $F$ can be approximated close to its minimum as a quadratic function of $\dot\theta^2$, $F \simeq F_0 + F_2(\dot\theta^2 - \dot\theta_0^2)^2$, where $F_0, F_2$ and $\dot\theta_0$ are constants [957]. The resulting approximate solution for $\theta$ is thus $\dot\theta \simeq \dot\theta_0$. The energy density of the field $F$, given by $\rho = (\dot\theta F' - F)/8\pi\tilde{G}$, then contains two terms: the first scales as $1/a^3$ (like matter), the second is constant (like vacuum energy). In this limit, the perturbations of the system have a small sound speed. An appropriate function $F$ away from the minimum is needed during the cosmological evolution.

▷ Linear fluctuations in cosmology behave as they would in normal general relativity in the presence of cold dark matter, but with an additional non-standard pressure that can be small enough [952]. In this way, the theory can reproduce the CMB temperature spectrum, an otherwise long-standing problem for the MOND idea (see the discussion on page 395). In the nonlinear regime it is expected that the two theories differ, but the observable consequences are yet to be explored.

▷ The introduction of a purely anti-symmetric $F_{\mu\nu}$, means that the waves in the symmetric $g_{\mu\nu}$ tensor can behave exactly as they do in the Einstein theory: expanding over a Minkowski background the photon and graviton speeds therefore remain equal [952].

The above set-up does not solve all the observational problems; open issues remain. For instance, it is not clear how the theory can explain the behaviour of the bullet cluster and of ordinary galaxy clusters, analogously to what discussed in sec. 11.3 [949]. The CMB polarization spectra might be affected by $U_\mu$. Furthermore, the action for $U_\mu$ seems not to be gauge invariant, endangering the quantum consistency of the theory [952].

## 11.4.2 MOND as a superfluid Dark Matter

As we saw above, tentative theories of MOND, when written in the universal language of Quantum Field Theory, employ new fields, similarly to theories of Dark Matter. One can similarly try to devise a theory of Dark Matter that acquires a MONDian behavior on galactic scales, while behaving as matter in cosmology. A possibility in this direction is that the scalar field $\theta$ has nothing to do with gravity, and mediates the desired MOND force because it couples to the matter density similarly to gravity, as $\theta\rho$. To match the MONDian dynamics, its non-relativistic Lagrangian must contain the unusual $|\boldsymbol{\nabla}\theta|^{3/2}$ term.

A possible justification for such a term could be DM super-fluidity [958]. Going against the original intention of MOND, let us assume the existence of a relativistic complex scalar field $\varphi$, which gives rise to the usual DM in cosmology. As discussed in section 3.4, DM occupation numbers become larger than unity in galaxies, if the DM mass is $M \lesssim \mathrm{eV}$. In this regime a phenomenon familiar from condensed-matter can take place: DM can form a Bose-Einstein condensate (as in section 3.4), and become super-fluid. This phenomenon occurs if the phonon fluctuations of the DM condensate are ultra-light. This composite degree of freedom can play the role of $\theta$, giving MONDian dynamics on galactic scales.

Indeed, the ultra-light phonon field $\theta$ can be understood in quantum field theory as the Goldstone boson of the global U(1) symmetry describing the conservation of the $\varphi$ particle number, which gets

spontaneously broken by the finite-density condensate $\langle\varphi\rangle = v\,e^{i\mu t}$, where $\mu$ is a chemical potential. Parameterizing the relativistic field as $\varphi(x) = [v + h(x)]e^{i[\mu t + \theta(x)]}$ and expanding its Lagrangian in the non-relativistic limit, the Goldstone boson $\theta$ acquires an effective Lagrangian of the desired MOND type, if the initial relativistic Lagrangian is

$$\mathscr{L}_{\text{MOND}}^{\text{super-fluid}} = |\partial_\mu\varphi|^2 - M^2|\varphi|^2 - V(\varphi) + \mathscr{L}_{\text{SM}}, \qquad (11.10)$$

with the unusual repulsive potential $V \propto c|\varphi|^6$ [958]. The effective action for the light phonon $\theta$ depends only on the combination $X \equiv \mu + \dot\theta - (\nabla\theta)^2/2M$, due to the Galilean invariance of the theory (i.e., due to the non-relativistic remnant of the full Lorentz invariance). The special form of the potential gives rise to the $X^{3/2}$ power needed to obtain MOND. However, MOND needs $c < 0$, which leads to instabilities. More complicated actions separate the $X = 0$ vacuum from such instabilities, and alternative models assume a non-minimal kinetic term. However issues such as super-luminality remain in the theories, questioning their fundamental viability [958]. Proceeding anyway, the phonon $\theta$ mediates a MOND-like force on galactic distances, if one adds to $\mathscr{L}_{\text{SM}}$ an additional U(1)-breaking interaction of the phonon to matter of the form $\theta\rho$. A very small coupling is needed to obtain the desired value of $a_* \sim H_0$. The end result is something akin to the Landau two-fluid model: a fraction of DM remains in the form of particles, while a fraction of it forms a superfluid with a cored galactic profile, which leads to the MONDian dynamics. In this context MOND is an outcome of special properties of the DM system, rather than an alternative to DM.

There are differences between the above set-up and the simple MOND postulate. For instance, it is reasonable to assume that, unlike gravity, phonons interact with baryon but not with photons. T. Mistele et al. (2023) in [958] then argue that this difference makes the super-fluid DM incompatible with the weak lensing data.

## 11.5   Conformal gravity

Finally, we summarise a different attempt at interpreting galactic rotation curves as modified gravity, without DM. In general, pure Einstein gravity, with the action $S_{\text{E}} = \int d^4x\sqrt{|\det g|}[-\bar{M}_{\text{Pl}}^2 R/2 - V]$, might be the low-energy limit of a deeper theory. An interesting possibility for the latter is the so called *conformal gravity* which has as the action

$$S_{\text{C}} = \int d^4x\sqrt{|\det g|}\frac{R^2/3 - R_{\mu\nu}^2}{f_2^2}. \qquad (11.11)$$

$S_{\text{C}}$ is non-trivially invariant under a Weyl rescaling of the metric, $g_{\mu\nu}(x) \to \Omega^2(x)g_{\mu\nu}(x)$ for arbitrary $\Omega(x)$. For example, $f_2$ is a dimension-less constant signalling scale invariance and renormalizability. Classical equations of conformal gravity admit a spherical solution [959]

$$ds^2 = f(r)\,dt^2 - \frac{dr^2}{f(r)} - r^2(d\theta^2 + \sin^2\theta\,d\varphi^2), \qquad f(r) = \sqrt{1 - 6GMk} - \frac{2GM}{r} + kr - \frac{V}{3\bar{M}_{\text{Pl}}^2}r^2, \ (11.12)$$

where $V$, $GM$ and $k$ are three constants of integration. One of them, $M$ corresponds to the local mass in the solutions to Einstein gravity. The two additional integration constants $k$ and $V$ arise because the conformal gravity action leads to equations of motion that contain 4 powers of derivatives and correspond to long-range effects. So they are expected to be the same in all different galaxies. In particular, in the $k = 0$ limit, eq. (11.12) reduces to the Schwarzschild-de Sitter vacuum solution of Einstein gravity with a vacuum energy $V$. In the small-$r$ limit eq. (11.12) reduces to Einstein gravity and to the Newton potential, in agreement with observations (for example in the solar system).

In the Newtonian limit, the extra $kr$ term corresponds to a gravitational potential growing with

distance, and thereby to a constant force. A constant force does not make galactic rotation curves flat (a force $F \propto 1/r$ would be needed). Nevertheless, it has been proposed that the observed galactic rotation curves can be reproduced by the appropriate form of $f(r)$, without the need for DM, by fitting different values of $M$ for each galaxy, while $k$ and $V$ are assumed to be universal [959].

This proposal faces several theoretical issues [959]: *i*) 4-derivative equations of motion imply the classical Ostrogradsky instability and a ghost state at quantum level. *ii*) A Weyl transformation and a redefinition of $r$ allows to remove the $kr$ term, bringing back eq. (11.12) to the Schwarzschild-de Sitter form. This issue is debated by Mannheim (2021) in [959]. *iii*) The conformal symmetry is broken at microscopic level, for example by the quantum running of SM couplings and by the Higgs mass. *iv*) It's unclear how such a proposal could give a DM-like effect in cosmological CMB and large scale structure data, and the associated gravitational lensing on cosmological scales. See [960] for a related attempt, not based on an action.

This underscores the many challenges faced by the efforts of trying to match some observations by only modifying gravity, while not introducing new degrees of freedom (such at DM).

# Appendix A

# Acronyms

For clarity, we list many TAA (Technical Acronyms and Abbreviations) employed in our review. We do not include names of experiments and collaborations, up to some exceptions.

ADM  Asymmetric Dark Matter.

AGN  Active Galactic Nucleus.

AU  Astronomical Unit.

BAO  Baryonic Acoustic Oscillations.

BBN  Big Bang Nucleosynthesis.

BH  Black Hole.

BR  Branching Ratio.

BSM  Beyond the SM.

CC  Cosmological Constant.

CDM  Cold Dark Matter.

CHAMP  CHArged Massive Particle (of DM).

CIB  Cosmic Infrared Background.

CL  Confidence Level.

CMB  Cosmic Microwave Background.

CMSSM  Constrained Minimal Supersymmetric Standard Model.

COB  Cosmic Optic Background.

CP  Charge Conjugation times Parity.

CPT  CP times Time reversal.

CR  Cosmic Rays.

DD  Direct Detection (of Dark Matter).

DE  Dark Energy.

DED  Dark Energy Domination.

DFSZ  Dine-Fischler-Srednicki-Zhitnitskii axion.

DIS  Deep Inelastic Scattering.

DM  Dark Matter.

DM$\nu$  Neutrinos from DM annihilations.

EBL  Extragalactic Background Light.

EFE  External Field Effect (in MOND).

EFT  Effective Field Theory.

dof  degree of freedom.

FIMP  Freeze-In Massive Particles.

FIMP  Feebly Interacting Massive Particles.

FLRW  Friedmann-Lemaître-Robertson-Walker (equations).

FS  Free Streaming.

GC  Galactic Center of the Milky Way.

GCH  Galactic Center Halo (a 'small' region around the GC).

GH  Galactic Halo.

GloC  Globular Cluster.

GR  General Relativity.

GRB    Gamma Ray Burst.

GUT    Grand Unified Theory.

GW     Gravitational Wave.

HDM    Hot Dark Matter.

HEP    High Energy Physics.

HS     Haloscope.

HST    Hubble Space Telescope.

IACT   Imaging Atmospheric Čerenkov Telescope.

IC     Inverse Compton.

ICS    Inverse Compton Scattering.

ID     Indirect Detection (of Dark Matter).

IDM    Inert Doublet Model/Inert Higgs.

IGM    InterGalactic Medium.

IMBH   Intermediate Mass Black Hole.

ISRF   InterStellar Radiation Field.

JWST   James Webb Space Telescope.

KSVZ   Kim-Shifman-Vainstein-Zakharov axion.

ΛCDM   The standard cosmological model featuring a CC and CDM.

ΛD     CC / DE Domination.

LEP    the most recent $e\bar{e}$ collider at CERN.

LHC    Large (or Last) Hadron Collider.

LKP    Lightest Kaluza-Klein Particle.

LSP    Lightest (Stable) SuperSymmetric Particle.

LSS    Large Scale Structure.

LSS    Last Scattering Surface.

LSW    Light Shining through a Wall (experiment).

MACHO  Massive Astrophysical Compact Halo Object.

MB     Maxwell-Boltzmann distribution.

MD     Matter Domination.

MDM    Minimal Dark Matter.

MOND   MOdified Newtonian Dynamics.

MSP    Milli-Second Pulsar.

MSSM   Minimal Supersymmetric Standard Model.

MReq   Matter-Radiation equality.

MW     Milky Way.

NFW    Navarro, Frenk and White DM density distribution.

NLSP   Next-to-Lightest SuperSymmetric Particle.

NRO    Non Relativistic Operator.

NRO    Non Renormalizable Operator.

NS/ns  Neutron Star.

PBH    Primordial Black Hole.

PDF    Parton Distribution Function.

PDF    Probability Distribution Function.

PIMP   Planckian Interacting Massive Particle.

pNGB   pseudo-Nambu-Goldstone boson.

PTA    Pulsar Timing Array.

QCD    Quantum Chromo Dynamics.

QED    Quantum Electro Dynamics.

QFT    (Relativistic) Quantum Field Theory.

QSO    Quasi-Stellar Object (Quasar).

RD     Radiation Domination.

SHM    Standard Halo Model.

SIDM   Self-Interacting Dark Matter.

SIMP   Strongly Interacting Massive Particle.

SIGW   Scalar Induced Gravitational Waves.

SM     Standard Model of particles.

SMBH   Super-Massive Black Hole.

SN     Supernova.

SNR    Supernova Remnant.

SSM    Standard Solar Model.

**SUSY** SUperSYmmetry.

**TeVeS** Tensor-Vector-Scalar theory.

**vev** vacuum expectation value.

**VIB** Virtual Internal Bremsstrahlung.

**WD/wd** White Dwarf.

**WDM** Warm Dark Matter.

**WIMP** Weakly Interacting Massive Particle.

**WMAP** Wilkinson Microwave Anisotropy Probe.

# Appendix B

# Basic notations and conventions

$a_0 = 1/\alpha m_e \approx 1/3.7\,\text{keV}$, Bohr radius.

$\alpha = e^2/4\pi \approx 1/137.036$, fine-structure constant.

$\alpha_s = g_s^2/4\pi$, strong coupling constant.

$\alpha_2 = g_2^2/4\pi$, weak $\text{SU}(2)_L$ coupling constant.

$\alpha_Y = g_Y^2/4\pi$, hypercharge coupling constant.

$d\eta$ : conformal time, $d\eta = dt/a$.

$\eta$: baryon-to-photon number density ratio, $\eta = n_b/n_\gamma$.

$\eta$: slow roll parameter in inflation.

$\epsilon$: slow roll parameter in inflation.

$\epsilon$: kinetic mixing.

$f_a$: axion decay constant.

$g_s$: strong gauge coupling.

$g_2$: weak $\text{SU}(2)_L$ gauge coupling.

$g_Y$: hypercharge gauge coupling.

$h$: Higgs boson

$h$: the scaling factor for Hubble expansion rate defined through $H_0 = h \times 100\,\text{km/sec}\cdot\text{Mpc}$.

$H$: Higgs doublet.

$H$: Hubble expansion rate.

$H_0$: present-day Hubble expansion rate.

$K$: kinetic energy.

$\Lambda_{\text{QCD}} \approx 300\,\text{MeV}$, non-perturbative QCD scale.

$M$: DM mass.

$M_h \approx 125\,\text{GeV}$, Higgs boson mass.

$m_N \approx 0.939\,\text{GeV}$, nucleon mass. We mostly work in the isospin limit $m_N = m_p = m_n$.

$m_a$: axion mass.

$m_n$: neutron mass.

$m_p$: proton mass.

$M_\odot \approx 1.988\,10^{30}\,\text{kg} = 1.115\,10^{57}\,\text{GeV}$, solar mass.

$M_\oplus \approx 5.972\,10^{24}\,\text{kg} = 3.350\,10^{51}\,\text{GeV}$, Earth mass.

$M_{\text{Pl}} \approx 1.2\,10^{19}\,\text{GeV}$, Planck mass.

$\bar{M}_{\text{Pl}}$: reduced Planck mass, $\bar{M}_{\text{Pl}}^2 = M_{\text{Pl}}^2/8\pi$.

$M_t \approx 171\,\text{GeV}$, top quark mass.

$n_b$: number density of baryons.

$n_\gamma$: number density of photons.

$\Omega_b$: cosmological abundance of normal baryonic matter, see eq. (C.10).

$\Omega_{\text{DM}}$: cosmological abundance of dark matter, see eq.s (1.1), (C.10).

$\wp$ : pressure.

$\text{Pf}$ : Pfaffian.

$s_\mathrm{W}$: $\sin\theta_\mathrm{W}$, sinus of the weak angle.

$\tau$: DM lifetime.

$T_\mathrm{eq} \approx 0.8\,\mathrm{eV}$ temperature at matter-radiation equality.

$T_\mathrm{U} \approx 13.7\,\mathrm{Gyr}$, the age of the Universe.

$T_0 \approx 2.726\,\mathrm{K}$, present CMB temperature.

$v \approx 174\,\mathrm{GeV}$, the Higgs vacuum expectation value.

$Y$: hypercharge, normalized such that the electric charge is $Q = T_3 + Y$.

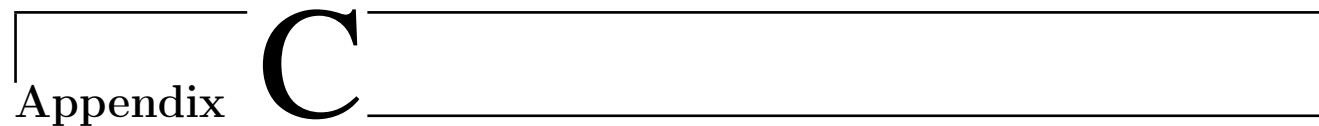

# Appendix C

# Cosmology

In this appendix we collect some of the main results from big-bang cosmology, which are used throughout the main text.

## C.1 Big bang expansion

The expanding Universe, assumed to be homogenous and isotropic (in agreement with observations, at least on large enough scales) is described by the translation and rotation invariant Friedmann-Lemaître-Robertson-Walker (FLRW) metric

$$ds^2 = -dt^2 + a^2(t) \left[ \frac{dr^2}{1 - k\, r^2} + r^2\, d\theta^2 + r^2 \sin^2 \theta\, d\phi^2 \right] \overset{\text{(if } k=0)}{=} -dt^2 + a^2(t)[dx^2 + dy^2 + dz^2], \quad \text{(C.1)}$$

where $a(t)$ is the scale factor and $k = 0, \pm 1$ is a constant that describes the global geometry ($k = 0$ for flat, $k = +1$ for closed, i.e., the geometry of a sphere, and $k = -1$ for open geometry, i.e., the geometry of a hyperbolic surface). The fact that $a$ depends on time encodes the expansion. In the usual convention, $a(t)$ is dimension-less and is normalized in such a way that it equals unity today, $a(t = T_U) = a_0 = 1$. Conventionally, one also defines the cosmological redshift $z$ as $1 + z = 1/a$, so that today $z_0 = 0$.

The matter-energy content of the Universe is modeled as a generic fluid, whose stress-energy tensor is constrained by the homogeneity and isotropy to have the form

$$T^\mu_\nu = \text{diag}\,(\rho, p, p, p), \quad \text{(C.2)}$$

where $\rho$ is the energy density and $p$ the pressure. In such a Universe, Einstein's equations of general relativity, $R_{\mu\nu} - \mathcal{R}g_{\mu\nu}/2 = 8\pi G\, T_{\mu\nu}$, specialize to the Friedmann equations

$$H^2 = \frac{8\pi G}{3}\rho - \frac{k}{a^2}, \qquad \frac{\ddot{a}}{a} = -\frac{4\pi G}{3}(\rho + 3p), \quad \text{(C.3)}$$

where the dot operator ( $\dot{}$ ) denotes the derivative with respect to time $t$. Here, $H \equiv \dot{a}/a$ is the *Hubble expansion rate*. Note that $H$ is not constant in time. In general, it varies as $H \propto 1/t$ in the phases of interest (see eq. (C.14) below). Its present value is the so called *Hubble 'constant'*, $H = H_0$, where

$$H_0 = h\frac{100\,\text{km}}{\text{sec} \cdot \text{Mpc}} = \frac{h}{9.78\ 10^9\,\text{yr}} = 2.1\,h\ 10^{-33}\,\text{eV}, \qquad h \approx 0.7, \quad \text{(C.4)}$$

with parsec equal to $1\,\text{pc} = 3.26\,\text{lyr}$.

Some of the equations simplify, if one uses instead of time $t$, the *conformal time* $\eta$, defined as the comoving distance traveled by light (we always set $c = 1$)

$$d\eta \equiv dt/a(t) = da/Ha^2. \tag{C.5}$$

It is fun to note that the first Friedmann equation in (C.3) could have been derived already by Newton (up to the 'Jeans swindle'), centuries before Einstein, Friedmann, Robertson and Walker. Indeed, modeling the Universe as a sphere with radius $a(t)$ composed by a constant density $\rho(t)$, the Newtonian gravity predicts that it evolves according to

$$\ddot{a} = -\frac{GM(r < a)}{a^2} = -\frac{4\pi G \rho(t)}{3} a. \tag{C.6}$$

Taking into account that for matter one has $\rho(t) \propto 1/a^3(t)$ this equation can be integrated, obtaining the usual conservation of energy

$$\frac{d}{dt}\left[\frac{1}{2}\dot{a}^2 - \frac{4\pi}{3}G\rho a^2\right] = 0,$$

from which eq. (C.3) follows.

The constant $k$ in (C.3) plays the role of an energy constant. The critical case of 'zero energy', $k = 0$, corresponds to a density that is equal to the 'critical density',

$$\rho = \rho_{\rm cr} \equiv 3H^2/8\pi G, \tag{C.7}$$

and describes a Universe that expands "for free" (the double of zero energy is zero energy). This is the case predicted by the inflationary cosmology in general relativity, and is compatible with the present observations of the actual Universe.

## C.2   Matter-energy content

Many components of the Universe are described by a simple equation of state

$$p = w\rho \qquad \text{with } w \text{ a constant.} \tag{C.8}$$

The relativistic conservation law for the energy momentum tensor, $T^{\mu\nu}{}_{;\nu} = 0$, gives the first law of thermodynamics:

$$dU = -p\,dV, \qquad U = \rho V. \tag{C.9}$$

The explicit solution is $\rho \propto a^{-3(1+w)}$. The relevant special cases are:

- $w = 0$ describes *non-relativistic matter* ('dust'), $p_m = 0$ and $\rho_m \propto 1/a^3$;

- $w = 1/3$ describes *relativistic particles* ('radiation'): $p_r = \rho_r/3$ and $\rho_r \propto 1/a^4$;

- $w = -1$ describes *vacuum energy*: $p_V = -\rho_V$ and $\rho_V \propto$ const,, with $a$.

The actual Universe is composed from a sum of different components: matter, radiation and dark energy. The dark energy contribution is at present observationally consistent with it being just the vacuum energy, and it might thus be described simply by a nonzero cosmological constant $\Lambda$, without any dynamics. The present energy densities of these components are conventionally expressed in terms of ratios with respect to the critical density

$$\Omega_i = \rho_i/\rho_{\rm cr}. \tag{C.10}$$

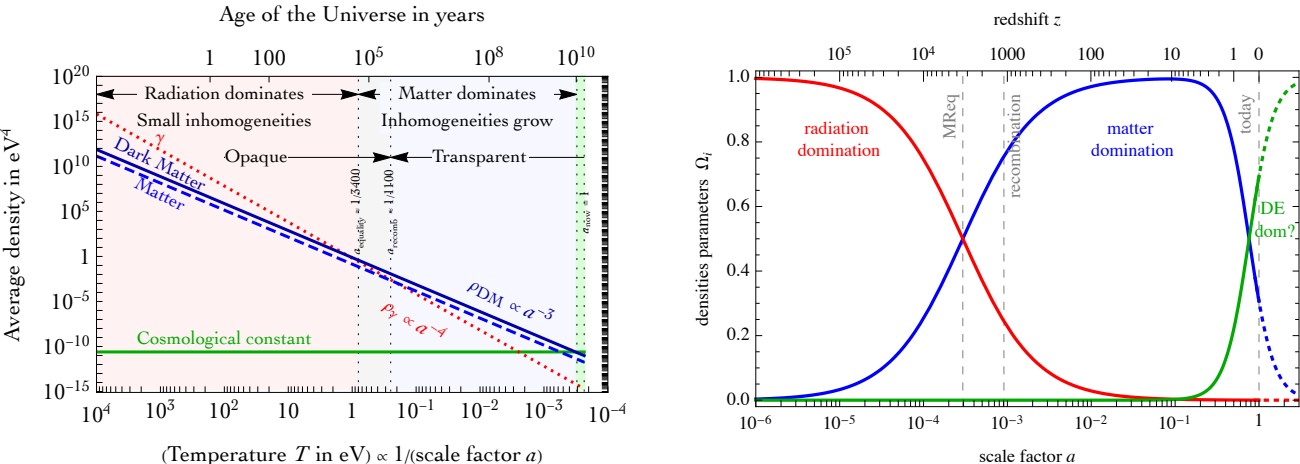

Figure C.1: **Evolution of the components of the (flat) Universe** as a function of time or temperature (left), and a simplified version showing the dominant component as a function of the scale factor a or the redshift z (right).

They are measured to be [3, 5]

$$\Omega_\Lambda \simeq 68.5\%, \qquad \Omega_m \simeq 4.9\% + 26.4\%, \qquad \Omega_r \simeq 0.0054\% + 0.0037\%, \qquad \sum_i \Omega_i \simeq 1. \qquad (C.11)$$

The matter contribution has two components: normal baryonic matter ($\Omega_b \simeq 4.8\%$) and the unknown Dark Matter ($\Omega_{DM} \simeq 25.8\%$), the subject of this review. Radiation is also made out of two components: photons (0.0054%, as per the energy density of a blackbody with temperature $T_{CMB} = 2.7255$ K) and neutrinos, here considered massless (0.0037%, as they contribute as 3.046 effective fermionic species, each with a temperature $(4/11)^{4/3} T_{CMB}$). Since the three components $\Omega_\Lambda, \Omega_m$, and $\Omega_r$, evolve differently with $a$, their relative weights in the composition of the Universe changes with time. From fig. C.1, we can see that matter started dominating the Universe at $a > a_{eq} = \Omega_r/\Omega_m \approx 1/3400$, i.e., at $t_{eq} \approx 60$ kyr (Matter-Radiation equality, MReq).

The age of the Universe is given by

$$T_U = \int_0^1 \frac{da}{a\,H} = \frac{1}{H_0}\int_0^1 \frac{da}{a\sqrt{\Omega_\Lambda + \Omega_m/a^3 + \Omega_r/a^4}} = \frac{1}{H_0} \times \begin{cases} 1/2 & \text{if } \Omega_r = 1, \\ 2/3 & \text{if } \Omega_m = 1, \\ 0.96 & \text{our Universe.} \end{cases} \qquad (C.12)$$

In general, one can solve numerically the Friedmann equation (C.3)

$$\frac{\dot{a}}{a} = H = H_0\sqrt{\frac{\rho}{\rho_0}} = H_0\sqrt{\Omega_\Lambda + \frac{\Omega_m}{a^3} + \frac{\Omega_r}{a^4}} \ . \qquad (C.13)$$

Simple analytic results hold for the epochs in which either matter (MD) or radiation (RD) dominated the total energy density,

$$a(t) = \begin{cases} (2tH_0)^{1/2} \\ (3tH_0/2)^{2/3} \end{cases} \qquad H(t) = \begin{cases} 1/(2t) & \text{during RD at } a \ll a_{eq}, \\ 2/(3t) & \text{during MD at } 1 \gg a \gg a_{eq}. \end{cases} \qquad (C.14)$$

# C.3   Particles in thermal equilibrium

A gas of relativistic particles at temperature $T$ has energy $E \sim T$, wavelength $\lambda \sim 1/T$, and consequently number density $n_{\text{eq}}$ and energy density $\rho_{\text{eq}}$ approximatively given by

$$n_{\text{eq}} \sim T^3, \qquad \rho_{\text{eq}} \sim T^4. \tag{C.15}$$

For non relativistic particles, $T \ll m$, one instead has $E \simeq m$ such that $\rho_{\text{eq}} = m n_{\text{eq}}$ and both densities get suppressed by a Boltzmann factor $e^{-m/T}$ (unless there is a conserved quantum number that prevents such suppression, e.g., the baryon number in the case of baryonic matter).

The precise formulæ for the thermal densities of a gas of particles with mass $m$ and $g$ degrees of freedom are:

$$n_{\text{eq}}(T) \;=\; g \int \frac{d^3 p}{(2\pi)^3} f_{\text{eq}} = \frac{g}{2\pi^2} \int_m^\infty f_{\text{eq}} E p \, dE, \tag{C.16}$$

$$\rho_{\text{eq}}(T) \;=\; g \int \frac{d^3 p}{(2\pi)^3} E f_{\text{eq}} = \frac{g}{2\pi^2} \int_m^\infty f_{\text{eq}} E^2 p \, dE, \tag{C.17}$$

$$p_{\text{eq}}(T) \;=\; g \int \frac{d^3 p}{(2\pi)^3} \frac{p^2}{3E} f_{\text{eq}} = \frac{g}{6\pi^2} \int_m^\infty f_{\text{eq}} p^3 \, dE, \tag{C.18}$$

where $f_{\text{eq}} = 1/(e^{E/T} + c)$ is the Fermi-Dirac distribution for fermions ($c = +1$) or the Bose-Einstein distribution for bosons ($c = -1$). A simple useful approximation is the Boltzmann limit $c = 0$.

▷ In the ultra-relativistic limit, $T \gg m$, the integrals can be performed easily, with the explicit results given by

$$n_{\text{eq}}(T \gg m) = \frac{g}{\pi^2} T^3 \times \begin{cases} 3\zeta(3)/4 & c = +1, \\ 1 & c = 0, \\ \zeta(3) & c = -1, \end{cases} \tag{C.19}$$

$$\rho_{\text{eq}}(T \gg m) = \frac{g}{\pi^2} T^4 \times \begin{cases} 7\pi^4/240 & c = +1, \\ 3 & c = 0, \\ \pi^4/30 & c = -1, \end{cases} \tag{C.20}$$

and $p_{\text{eq}} = \rho_{\text{eq}}/3$. The total energy density in relativistic species, which can have different temperatures, is then given by

$$\rho_{\text{R}}(T) = \frac{\pi^2}{30} g_\rho(T) \, T^4, \qquad \text{with} \quad g_\rho(T) = \sum_{m \ll T} g_{\text{b}} \left( \frac{T_{\text{b}}}{T} \right)^4 + \frac{7}{8} \sum_{m \ll T} g_{\text{f}} \left( \frac{T_{\text{f}}}{T} \right)^4, \tag{C.21}$$

where the two sums run over bosons and fermions, respectively.

▷ In the non-relativistic limit, $T \ll m$, all the distributions reduce to the Boltzmann result,

$$n_{\text{eq}}(T \ll m) = g(mT/2\pi)^{3/2} e^{-m/T}, \qquad p_{\text{eq}} \ll \rho_{\text{eq}} \simeq m n_{\text{eq}}. \tag{C.22}$$

▷ For generic $T$, one can obtain analytic expressions by approximating the quantum distributions with a Boltzmann distribution

$$n_{\text{eq}} \approx \frac{g m^2 T}{2\pi^2} \text{K}_2 \left( \frac{m}{T} \right), \qquad \rho_{\text{eq}} \approx \frac{g m^2 T}{2\pi^2} \left[ m \, \text{K}_1 \left( \frac{m}{T} \right) + 3T \, \text{K}_2 \left( \frac{m}{T} \right) \right], \tag{C.23}$$

where $K_{1,2}(x)$ denote the modified Bessel functions of the second kind.

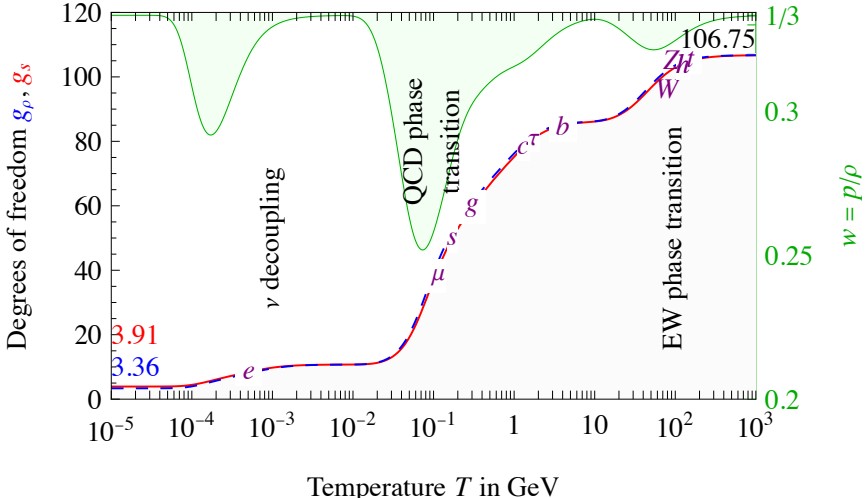

Figure C.2: *Degrees of freedom of the SM $g_s(T)$ (red) and $g_\rho(T)$ (blue dashed), increasing up to 106.75 as function of the temperature $T$. The green curve (with values on the right axis) shows the approximate value of $w = p/\rho$ (see [961] for lattice computations).*

Since the Universe cannot exchange the heat with any "outside" system, its evolution in the thermal equilibrium conserves the entropy, $S = sV$, where $V$ is the volume, and

$$s = \frac{\rho + p}{T} \equiv \frac{2\pi^2}{45} g_s(T) \, T^3, \qquad \text{is the entropy density.} \tag{C.24}$$

Up to factors of order one, $s$ is essentially the total number density of particles, $n_{\rm eq}$. Conservation of $S$ implies that the temperature $T$ decreases during expansion as $T \propto 1/a$, up to the decrease in the effective number of (entropy density) degrees of freedom, $g_s$. The latter takes place whenever the various SM particles become non relativistic and can no longer be produced by thermal processes in the plasma. The entropy is dominated by the ultra-relativistic particles. Summing over all the relativistic degrees of freedom one thus has

$$g_s = \sum_{m \ll T} g_{\rm b} \left( \frac{T_{\rm b}}{T} \right)^3 + \frac{7}{8} \sum_{m \ll T} g_{\rm f} \left( \frac{T_{\rm f}}{T} \right)^3, \tag{C.25}$$

where $g_{\rm b,f}$ give the internal number of degrees of freedom (spin or any other internal quantum numbers) for each species of bosons and fermions. At high temperatures, all the SM fermions and bosons are in a thermal equilibrium and consequently have the same temperature $T$: the value of $g_s(T)$ in the SM is shown in fig. C.2. The effective number of degrees of freedom relevant for energy density, $g_\rho$, is defined analogously, but with $(T_{\rm b,f}/T)^4$ temperature ratios. At $T \gg M_t$, such that all SM particles are ultra-relativistic, one has $g_s = \frac{7}{8} 2 \cdot 3 \cdot 15 + 2(1 + 3 + 8) + 2 \cdot 2 = 106.75$, slightly lower than 118, the total number of SM degrees of freedom. This then slowly drops when different species of SM particles become non-relativistic. For $T \sim \Lambda_{\rm QCD}$ the calculation of $g_s$ is more complicated, see [961] for details. After neutrino decoupling, at $T \ll m_e$, the Universe contains photons and neutrinos with different temperatures, $T_\nu = T_\gamma (4/11)^{1/3}$. Measuring the entropy in units of the photon temperature, one has $g_s = 2 + \frac{7}{8} 2 \cdot 3 \frac{4}{11} = \frac{43}{11} \simeq 3.91$.

Additional light degrees of freedom that could be present in plasma are usually expressed in terms of the corresponding change of the effective number of neutrinos, $\Delta N_{\rm eff}$. The SM value of $N_{\rm eff}^{\rm SM}$ is defined

via the low temperature ratio of the neutrino energy, $\rho_\nu$, to the photon energy density, $\rho_\gamma$,

$$\left.\frac{\rho_\nu}{\rho_\gamma}\right|_{T/m_e \to 0} = \frac{7}{8}\left(\frac{4}{11}\right)^{4/3} N_{\text{eff}}^{\text{SM}}, \tag{C.26}$$

and is $N_{\text{eff}}^{\text{SM}} = 3.044$ (it differs from 3 due to subleading effects such as the remaining small energy transport from photon plasma to the neutrino sector). The contribution of any new relativistic degrees of freedom to the energy density of plasma can then be parametrized similarly via

$$\left.\frac{\rho_{\text{light NP}}}{\rho_\gamma}\right|_{T/m_e \to 0} = \frac{7}{8}\left(\frac{4}{11}\right)^{4/3} \Delta N_{\text{eff}}. \tag{C.27}$$

Current cosmological bounds require $\Delta N_{\text{eff}} \lesssim 0.3$.

## C.4   Particles out of thermal equilibrium

If the system is in a thermal equilibrium, the description of its physical properties is rather simples since everything is dictated by the temperature $T$, and possibly by the values of conserved quantum numbers such as the baryon number.

However, during the cosmological evolution particles may go out of thermal equilibrium. This is a potentially important effect for DM, since it can results in a relic abundance of DM. The criterion for the out of equilibrium dynamics can be succinctly stated as: *a particle is in thermal equilibrium, if $\Gamma \gg H$, i.e., if its interaction rate $\Gamma \sim n\sigma$ is faster than the rate of expansion, $H \sim T^2/M_{\text{Pl}}$.* The cross section $\sigma$ is determined by particle physics, where the interactions can be split into two broad categories:

- **Renormalizable interactions**, whose strength is controlled by dimension-less couplings $g$. Then $\sigma \sim g^{2-4}/T^2$ and $\Gamma \sim g^{2-4}T$ where the powers of 2 and 4 are, respectively, valid for $1 \to 2$ and $2 \to 2$ scatterings. Since $H$ drops quadratically with smaller $T$, these interactions become *fast* (relative to $H$) at late times, when the temperature falls below $T \lesssim g^{2-4}M_{\text{Pl}}$. For example, the electromagnetic, weak, and strong interactions have couplings $e, g_2, g_3 \sim 1$, and are in thermal equilibrium for temperatures below $T \sim 10^{17}$ GeV. The Yukawa interactions of charged SM particles span 5 orders of magnitude. The smallest one is the electron Yukawa coupling $\lambda_e \sim 10^{-6}$. The interactions mediated by it enter in thermal equilibrium at $T \sim$ TeV (this is of mainly academic interest since other interactions, namely the exchanges of gauge bosons already keep electrons and other SM particles in thermal equilibrium).

- **Non-renormalizable interactions**, whose strength is suppressed by powers of high scale, for instance, for dimension 6 interactions the dimensionful couplings are $\sim 1/M^2$. Then, for dimensional reasons, one has $\sigma \sim T^2/M^4$, $\Gamma \sim T^5/M^4$, so that such interactions become *slow* at late times $T \lesssim M(M/M_{\text{Pl}})^{1/3}$. For example, weak interactions at $T \ll M_{W,Z}$ are suppressed by $M \approx v$ and decouple at $T \sim$ few MeV. Electromagnetic interactions of electrons at $T \ll m_e$ are described by $M \approx m_e$, allowing for photon decoupling as $\Gamma \propto T^3\sigma_{\text{T}} \sim T^3/m_e^2$, where $\sigma_T$ is the Thomson cross section (although the formation of neutral hydrogen occurs earlier).

Particles that fall out of thermal equilibrium, i.e., particles with negligible interactions are easily described: their wavelength expands with universe, such that $p \propto 1/a$. Then:

- **For relativistic particles** (such as neutrinos) this means $E = p \propto 1/a$: they maintain a thermal distribution with $T \propto 1/a$, just as if they had remained in thermal equilibrium.[1]

---

[1]Note, however, that the temperature of the thermal plasma and the decoupled particles need not be the same. The temperature evolution for thermal plasma does not always follow $T \propto 1/a$ since the number of degrees $g_{\text{s}}(T)$ may change as the plasma cools.

- **For non-relativistic particles** (such as possibly DM) this means that $E = p^2/2m \propto 1/a^2$: they cool faster than particles in thermal equilibrium.

In summary:

|  | massless, $T \gg m$ | massive, $T \ll m$ |
|---|---|---|
| interacting with the plasma | $T \propto 1/a$ | $T \propto 1/a$ |
| non-interacting with the plasma | $T \propto 1/a$ | $T \propto 1/a^2$ |

For example, electrons become non relativistic at $T \sim m_e$, at which point the neutrinos have already decoupled. Neutrinos remain relativistic for a long time, but are non-relativistic at present (at least the two species that are required to be massive by the neutrino oscillation measurements). The kinetic energy of DM particles after decoupling tends to be red-shifted away.

## C.5   Boltzmann equations

Many important computations in cosmology are performed through the use of Boltzmann equations. Below, we review this useful tool.

In the absence of interactions the number of particles in a comoving volume $V$ remains constant. Boltzmann equations then encode the effects of different interactions. Let us study, e.g., how the decays, $1 \leftrightarrow 2 + 3$, and the inverse decays, $2 + 3 \to 1$, affect the number of particles of species '1', given by the number density $n_1$ in volume $V$,

$$
\frac{d}{dt}(n_1 V) = V \int d\boldsymbol{p}_1 \int d\boldsymbol{p}_2 \int d\boldsymbol{p}_3 \, (2\pi)^4 \delta^4(p_1 - p_2 - p_3) \times \tag{C.28}
$$
$$
\times [-|\mathscr{A}_{1\to 23}|^2 f_1(1 \pm f_2)(1 \pm f_3) + |\mathscr{A}_{23\to 1}|^2 (1 \pm f_1) f_2 f_3],
$$

where $d\boldsymbol{p}_i = \sum d^3 p_i / 2E_i (2\pi)^3$ is the relativistic phase space of particle $i$ summed over the $g_i$ degrees of freedom (polarizations and multiplet components); $|\mathscr{A}_{1\to 23}|^2$ and $|\mathscr{A}_{23\to 1}|^2$ are the squared transition amplitudes; $f_i$ are the energy distributions of the various particles, where the plus (negative) signs in $(1 \pm f_i)$ factors apply for $i$ particle that is a boson (fermion). For simplicity we assume that CP violation can be neglected, so that the decay and the inverse decay process have a common amplitude $\mathscr{A}$.

In principle, one should study the evolution of all $f_i$ in order to obtain the total number densities $n_i = \sum \int f_i \, d^3 p / (2\pi)^3$. In practice, the elastic scatterings (i.e., the interactions that do not change the number of particles and their species) are typically fast enough that they maintain *kinetic equilibrium*, so that the full Boltzmann equations for $f$ are solved by $f(p) = f_{\rm eq}(p) n / n_{\rm eq}$ where $f_{\rm eq} = [e^{E/T} \pm 1]^{-1}$ are the thermal equilibrium Bose-Einstein and Fermi-Dirac distributions. In this limit, each particle species is simply characterized by its total abundance $n$, that can be varied only by inelastic processes.

The factors $1 \pm f_i$ in eq. (C.28) take into account Pauli-blocking (for fermions) and stimulated emission (for bosons). Since the average particle energy is $\langle E \rangle \sim 3T$ one can within 10% accuracy set $1 \pm f \approx 1$, one thus approximate $f_{\rm eq}$ with the Boltzmann distribution $f_{\rm eq} \approx e^{-E/T}$. This approximation then leads to significant simplifications in the results.

When the inelastic processes are also sufficiently fast to maintain *chemical equilibrium*, the total number $n_{\rm eq}$ of particles with mass $M$ at temperature $T$ are given by

$$
n_{\rm eq} = g \int \frac{d^3 p \, f_{\rm eq}}{(2\pi\hbar)^3} = \frac{g M^2 T}{2\pi^2} \mathrm{K}_2 \left( \frac{M}{T} \right) = \begin{cases} g T^3 / \pi^2, & T \gg M, \\ g(MT/2\pi)^{3/2} e^{-M/T}, & T \ll M, \end{cases} \tag{C.29}
$$

where $g$ is the number of spin, gauge, etc, degrees of freedom. The factor $\hbar = h/2\pi$ has been explicitly shown to clarify the physical origin of the $2\pi$ in the denominator.

The Boltzmann equation for $n_1$ then simplifies to

$$\frac{1}{V}\frac{d}{dt}(n_1 V) = \int d\boldsymbol{p}_1 \int d\boldsymbol{p}_2 \int d\boldsymbol{p}_3\, (2\pi)^4 \delta^4(p_1 - p_2 - p_3) \times \tag{C.30}$$

$$\times |\mathscr{A}|^2 \left[ -\frac{n_1}{n_1^{\text{eq}}} e^{-E_1/T} + \frac{n_2}{n_2^{\text{eq}}} \frac{n_3}{n_3^{\text{eq}}} e^{-E_2/T} e^{-E_3/T} \right].$$

One can recognize that the integrals over final-state momenta reconstruct the decay rate $\Gamma_1$, and that the integral over $d^3 p_1 / E_1$ averages it over the thermal distribution of initial state particles. The factor $1/E_1$ corresponds to the Lorentz dilatation of the lifetime. The final result is therefore,

$$\frac{1}{V}\frac{d}{dt}(n_1 V) = \langle \Gamma_1 \rangle n_1^{\text{eq}} \left[ \frac{n_1}{n_1^{\text{eq}}} - \frac{n_2}{n_2^{\text{eq}}} \frac{n_3}{n_3^{\text{eq}}} \right], \tag{C.31}$$

where $\langle \Gamma_1 \rangle$ is the thermal average of the Lorentz-dilatated decay width

$$\Gamma_1(E_1) = \frac{1}{2E_1} \int d\boldsymbol{p}_2\, d\boldsymbol{p}_3 (2\pi)^4 \delta^4(p_1 - p_2 - p_3) |\mathscr{A}|^2. \tag{C.32}$$

If $\langle \Gamma_1 \rangle \gg H$, then the l.h.s. is much smaller than $\langle \Gamma_1 \rangle n_1^{\text{eq}}$ and thus the term in square brackets in eq. (C.31) is forced to be vanishingly small. This just means that there are inelastic interactions much faster than the expansion rate, which are able to enforce the chemical equilibrium, giving $n = n_{\text{eq}}$.

Let us apply this to a concrete example, the $N_1 \to LH$ decays in leptogenesis. The $2, 3 = L, H$ particles have fast gauge interactions. Therefore we do not have to write and solve Boltzmann equations for $L, H$, because we already know their solution: $L, H$ are kept in chemical thermal equilibrium. We only need to insert this result in the Boltzmann equation for $N_1$, which then simplifies to

$$\dot{n}_1 + 3Hn_1 = \langle \Gamma_1 \rangle (n_1 - n_1^{\text{eq}}), \tag{C.33}$$

having used $\dot{V}/V = 3H = -\dot{s}/s$.

In the numerical evaluations in computer codes one prefers to avoid very large or very small numbers. To achieve this it is convenient to reabsorb the $3H$ term, which accounts for the dilution due to the overall expansion of the universe, by normalizing the number density $n$ to the entropy density $s$. Therefore, we study the evolution of $Y = n/s$, as function of $z = m_N/T$, in place of time $t$. Here, $H\, dt = d\ln R = d\ln z$, since during the adiabatic expansion $sV$ is constant, i.e., $V \propto 1/T^3$. Expressed in terms of $Y(z)$ variables, the general form of the Boltzmann equations is

$$sHz\frac{dY_1}{dz} = \sum \Delta_1 \cdot \gamma_{\text{eq}}(12\cdots \leftrightarrow 34\cdots) \left[ \frac{Y_1}{Y_1^{\text{eq}}} \frac{Y_2}{Y_2^{\text{eq}}} \cdots - \frac{Y_3}{Y_3^{\text{eq}}} \frac{Y_4}{Y_4^{\text{eq}}} \cdots \right], \tag{C.34}$$

where the sum runs over all the processes that change the number of '1' particles by $\Delta_1$ units (e.g., $\Delta_1 = -1$ for the $1 \to 23$ decay, while $\Delta_1 = -2$ for the $11 \to 23$ scatterings, etc.). The quantity $\gamma_{\text{eq}}$ is the space-time density of interactions (i.e. the number of interactions per unit volume and unit time) in thermal equilibrium for the various processes.

The direct and inverse processes have the same reaction densities, as long as CP-violating effects can be neglected. The explicit expressions for 1-body and 2-body processes are then, as follows:

- For a **decay** and for its inverse process one gets the previous result,

$$\gamma_{\text{eq}}(1 \to 23) = S_f \sum_{\text{all}} \int d\boldsymbol{p}_1\, f_1^{\text{eq}} \int d\boldsymbol{p}_2\, d\boldsymbol{p}_3\, (2\pi)^4 \delta^4(p_1 - p_2 - p_3)|\mathscr{A}|^2 = \gamma_{\text{eq}}(23 \to 1), \tag{C.35}$$

where the sum runs over all the final-state and initial-state indices (polarisations, components of

gauge multiplets, etc), and $S_f = 1/n_f!$ is a symmetry factor, which differes from one, if the final state contains $n_f$ identical particles. The thermal average of the decay rate can be analytically computed in terms of Bessel functions,

$$\gamma_{\mathrm{eq}}(1 \to 23\cdots) = \gamma_{\mathrm{eq}}(23\cdots \to 1) = n_1^{\mathrm{eq}}\frac{K_1(z)}{K_2(z)}\Gamma(1 \to 23\cdots) \tag{C.36}$$

with $K_1(z)/K_2(z) \simeq 1$ in the non-relativistic limit $z \gg 1$.

- For a **2-body scattering process**

$$\gamma_{\mathrm{eq}}(12 \leftrightarrow 34) = S_i S_f \sum_{\mathrm{all}} \int d\boldsymbol{p}_1 \, d\boldsymbol{p}_2 \, f_1^{\mathrm{eq}} f_2^{\mathrm{eq}} \int d\boldsymbol{p}_3 \, d\boldsymbol{p}_4 \, (2\pi)^4 \delta^4(p_1 + p_2 - p_3 - p_4)|\mathscr{A}|^2, \tag{C.37}$$

where

$$S_i = \begin{cases} 1 & \text{if particles 1 and 2 are different,} \\ 1/2 & \text{if particles 1 and 2 are identical,} \end{cases}$$

is an initial-state symmetry factor analogous to $S_f$. One can perform analytically almost all of the integrals, and obtain

$$\gamma_{\mathrm{eq}}(12 \to 34\cdots) = \frac{T}{64\pi^4}\int_{s_{\mathrm{min}}}^{\infty} ds \, \sqrt{s} \, \hat{\sigma}(s) \, K_1\left(\frac{\sqrt{s}}{T}\right), \tag{C.38}$$

where $s = (p_1 + p_2)^2$ and $t = (p_1 - p_3)^2$ are the usual kinematical variables and the dimension-less 'reduced cross section' $\hat{\sigma}$ is defined as

$$\hat{\sigma} = S_i S_f \sum_{\mathrm{all}} \int_{t_{\mathrm{min}}}^{t_{\mathrm{max}}} dt \frac{|\mathscr{A}|^2}{8\pi s}. \tag{C.39}$$

The sum is over the initial *and* the final state indices.

One can express $\hat{\sigma}$ in terms of the usual cross section

$$\hat{\sigma} = \sigma \times 2sS_i g_1 g_2 \lambda, \tag{C.40}$$

where $\sigma$ is summed over the final state, but *averaged* over the initial state indices,

$$\sigma = \frac{S_f}{2E_1 2E_2 v}\sum_{\mathrm{final}} \int d\boldsymbol{p}_3 \, d\boldsymbol{p}_4 \, (2\pi)^4 \delta^4(p_1 + p_2 - p_3 - p_4)\langle|\mathscr{A}|^2\rangle_{\mathrm{initial}} = S_f \sum_{\mathrm{final}} \int dt \frac{\langle|\mathscr{A}|^2\rangle_{\mathrm{initial}}}{16\pi s^2 \lambda}. \tag{C.41}$$

The denominator, $E_1 E_2 v = [(p_1 \cdot p_2)^2 - m_1^2 m_2^2]^{1/2} = s\sqrt{\lambda}/2$ is a relativistic invariant. Here $\lambda \equiv [1 - (m_1 + m_2)^2/s][1 - (m_1 - m_2)^2/s]$ and $v^2 = (\boldsymbol{v}_1 - \boldsymbol{v}_2)^2 - (\boldsymbol{v}_1 \times \boldsymbol{v}_2)^2$ is the relative velocity. The result for the interaction rate is

$$\gamma_{\mathrm{eq}}(12 \to 34\cdots) = S_i g_1 g_2 \frac{T}{32\pi^4}\int_{s_{\mathrm{min}}}^{\infty} ds \, s^{3/2} \, \lambda \, \sigma \, K_1\left(\frac{\sqrt{s}}{T}\right). \tag{C.42}$$

In the non-relativistic limit, relevant for the DM freeze-out, the cross section times the relative velocity approaches a constant for $s$-wave annihilation

$$\lim_{v \to 0}\langle\sigma v\rangle = \sigma_0 \tag{C.43}$$

where $\sigma_0$ is the $s$-wave cross section. Taking into account that $v \simeq 2\sqrt{\lambda}$ and assuming that the 1 and

2 initial-state particle are those that can scatter non-relativistically, the reduced cross section and the interaction rate become

$$\hat{\sigma} \stackrel{T \ll m}{\simeq} s g_1 g_2 \sqrt{\lambda} S_i \sigma_0, \qquad \gamma_{\text{eq}} \stackrel{T \ll m}{\simeq} S_i n_1^{\text{eq}} n_2^{\text{eq}} \sigma_0. \tag{C.44}$$

The latter equation follows trivially from eq. (C.38). The text below eq. (4.13) discusses how to deal with the symmetry factor $S_i$.

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

# Index