# Peer review of "Dark Matter"

_SciPost Physics Reviews_

## Round 1 · Referee Report · Anonymous (Referee 1) · 2025-7-17

Report

This review is very timely and summarizes many important progresses in the past decades, from dark matter evidence, theoretical models, production mechanisms, detection methods and even anomalies. It would provide a comprehensive overview of the field and serve as a quite useful reference. I highly recommend it for publication.

There are some suggestions which I shall leave to the authors to decide whether they update or not. Main of the detection methods are described for particle dark matter in details. However in recent ten years, there have been increasing discussions about ultralight dark matter and primordial black holes, which although reviewed but very briefly. In particular, gravitational wave observations, like PTA, LIGO, even future LISA, may provide interesting probes.

A comment about the cold/hot dark matter in Sec. 3.3 at page 76, "DM must be cold or at least not too hot. Technically, this means that DM needs to be non-
relativistic at the time of matter-radiation equality". I think DM should also be non-relativistic below keV, much before matter-radiation equality, otherwise the matter power spectrum would be different.

Recommendation

Publish (meets expectations and criteria for this Journal)

  • validity: -
  • significance: -
  • originality: -
  • clarity: -
  • formatting: -
  • grammar: -

Author:  Marco Cirelli  on 2025-11-02  [id 5983]

(in reply to Report 1 on 2025-07-17)

We thank Referee 1 very much for their comments in Report #1. We address them here below and we have implemented the changes in the latest version of the manuscript.

Concerning the first comment (about the discussion of ultralight dark matter and primordial black holes as well as GW observations), we would like to point out that our manuscript does include several sections on those candidates and on the detection methods. However, they are not necessarily in Part II (the one dedicated to DM detection) and therefore maybe more difficult to spot. We apologie for that, but couldn’t find a better arrangement. We have now added some pointers and internal references to the relevant sections. Some of these are: - Section 3.1.1, discussing Primordial black holes and their detection methods. - Section 3.4.2, discussing ultra-light DM. - Section 5.8, discussing laboratory searches of light and ultra-light DM. - Section 6.14, discussing the galactic and stellar probes of ultra-light DM. - Section 10.4.5, discussing the searches for axion DM, including axion-like very light DM particles. We have also now added, in section 10.4.5, a new bullet item and a new citation block on gravitational wave experiments that allow to probe axions and other ultra-light DM candidates.

Comment about the cold/hot dark matter in Sec. 3.3: We have modified the sentence to make it clearer that the condition applies at the time that structures start to form, somewhat before matter-radiation equality. The keV scale is discussed in the rest of the text and in sec. 3.3.1.

---

## Round 1 · Referee Report · Anonymous (Referee 2) · 2025-8-26

Strengths

  1. Lucid.
  2. Comprehensive.
  3. Well-written.
  4. Well structured.
  5. Extensive bibliography.

Weaknesses

No weaknesses.

Report

My approach to this reviewing task was to fully read:

(a) sections on which I am an expert, to see if that narrative is technically correct, and
(b) sections I am less expert on, to see if I could understand them and learn from them.

As per the editor’s instructions, and given the length of the review article, I did not read the whole thing in detail. However, I did at least skim the sections I did not read in full.

In inverse relation to the number of pages in the review, which the authors note is roughly equal to the log of the number of string vacua, my report will be fairly short.

In essence, despite the exhaustive length, the quality of the exposition is very high, and consistently so. All of the explanations and summaries I looked at in detail were interesting to read, and surprisingly easy to understand. And, yes, the sections on which I am an expert are accurate, and the sections on which I am not an expert I also found easy to understand and I did learn from them.

The authors are, of course, to be commended on their encyclopaedic thoroughness. No topics of any currency were neglected, including modified gravity theories. The choices they made about how detailed to become (e.g. not trying to cover every single dark matter model ever proposed) are reasonable, and in fact their curation of subject matter is a major reason behind the readability of the paper. The extensive, though not actually exhaustive, references will be an excellent general resource.

As the authors note, the non-gravitational discovery of dark matter, or even a dark matter candidate, will render a fair amount of the material irrelevant in a narrow sense, although the theoretical insights gained through the various “thought experiments” about what the dark matter may be should have some lasting value. But it is a reasonable bet, though I hope I am wrong, that this kind of observational/experimental revelation is still quite some time off. I imagine this review will be a standard reference for many years.

I recommend publication without any revision.

Requested changes

  1. I did find the occasional misspelling (i.e. typo) which hopefully can be dealt with by a spell check during production.

Attachment

Recommendation

Publish (surpasses expectations and criteria for this Journal; among top 10%)

---

## Round 1 · Referee Report · Anonymous (Referee 3) · 2025-8-27

Strengths

An outstanding and up-to-date review of a frontier research field,.
Combines theoretical and experimental status.
A comprehensive review with outstanding clarity and accessible to both the astrophysics and particle physics communities, a truly rare achievement in this interdisciplinary field.

Weaknesses

Section 1.1.3
DM deficient dwarf galaxy explanations should be better explained. Specifically the bullet hypothesis for such objects has recently (2025) received strong support from van Dokkum et al. I don't necessarily expect the authors to include this reference as it one of a number of relevant references that are very recent, and its clear the authors rightly need to draw a line on updates. However the cited discussion (Sec 1.2.1) of the DM deficient dwarf galaxies in subsection 1.1.3 is misleading as it refers to the Bullet cluster. Other corrections are suggested below with more specific comments.

Report

Excellent topical and accessible review of a core subject in cosmology that does full justice to combining theoretical and experimental issues.

Requested changes

This review merits a detailed abstract. Much more than one sentence please.

The following comments are based on my reading in detail of the first few main sections.

1.3.1; also 1.3.4
A nice review of the formation of structure formation in the standard CDM model from an analytic perspective.
What is missing are comments on the late-time role of Lambda which is now part of the standard model.

1.3.5
Since "the bulk of D and 4He and a good portion of 3He and 7Li" are mentioned here, whatever this means, there should be specific references to possible unresolved BBN anomalies.

Ch 2
"DM tends to be roughly spherically distributed in gravitationally bound systems'". How confident are we of this? More technical discussion is given below of course but this does seem to me to be a strong assertion.

2.1.2
"the classical theory of collapse of isolated, spherical, collision-less and dissipation-less systems predicts equilibrium DM halos that are singular isothermal spheres."
What about initial spin or rotation, admittedly for non-isolated systems? Some citation of works by Fall et al might be appropriate, eg just refer to 2.4.2

Recommendation

Ask for minor revision

  • validity: top
  • significance: top
  • originality: high
  • clarity: top
  • formatting: excellent
  • grammar: excellent

Author:  Marco Cirelli  on 2025-11-02  [id 5982]

(in reply to Report 4 on 2025-08-27)

We thank Referee 3 for the very pertinent remarks of their Report #4. We respond here below, and we have implemented the corresponding changes in the latest version of the manuscript.

Section 1.1.3: "DM deficient galaxies":
We thank the referee for point out the recent results by Keim, van Dokkum at al. (2025). We assume they are referring to 2506.10220. We have now updated the discussion by pointing to this reference, and corrected the misleading impression of referring to the bullet cluster.

Abstract:
we thank the referee for the suggestion but we would like to stick to one sentence. Including more information in the abstract would essentially mean replicating the table of contents, or the Index located at the end of the manuscript. We trust that the readers can find the information they need even without a detailed abstract.

1.3.1: "late-time role of Lambda":
We thank the referee for pointing this out. While a thorough review of the role of Dark Energy in structure formation is out of our scope, we do have some side comments in the text concerning this point. In sec. 1.3.1, we point out that clustering slows down in DE-domination. In sec. 1.3.2 we mention numerical simulations that include DE. We have now added a whole block of references (item [24], “Dark Energy in structure formation and in numerical simulations”) that address this.

1.3.5: "BBN anomalies":
Such BBN anomalies, with respect to the possible role of DM in them, are briefly discussed in section 6.12 “DM and Big Bang Nucleosynthesis”. Following the referee’s suggestion, we added a pointer to section 6.12 from section 1.3.5. We also specified that 3He and 7Li are affected by subsequent astrophysical processes.

Ch 2: "DM spherically distributed":
A detailed discussion of the deviations from spherical symmetry is offered in section 2.4. We have therefore added a pointer to that section.

2.1.2: "initial spin or rotation in collapse theory":
We added the citation to the classical work of S.M. Fall and G. Efsfathiou (1980), in the citation block related to sec. 2.4.2, as suggested.

---

## Round 1 · Referee Report · Anonymous (Referee 4) · 2025-8-29

Strengths

1- Most ambitious attempt ever made to review the vast and fast-evolving topic of dark matter. In my opinion, largely successful.

2- Clear both in the organisation and in the presentation.

3- Original in the organisation and presentation of existing knowledge on variety of topics.

4- Complete

5- Insightful

Report

I have reviewed this submission by reading all Chapters, but I went more into details of Chapter 1 to 5 and of a few sections of the other Chapters. In doing so, I abided by the editor's invitation to referee only several chapters.

The submission presents a review of dark matter, object of enormous past and ongoing interest in physics. As far as I can judge, the presentation is correct, up-to-date and complete, thus succeeding in a highly non-trivial challenge given the vastness of the literature on the topic. The submission succeeds also in presenting both known and speculative facts about dark matter in a way that is logically clear, systematic, and insightful. This makes it easy to navigate and useful for different audiences, going from master students to expert researchers. In these respects and as an example, I find particularly useful Tables 2.2, 2.3 5.1, 5.4, 6.2, 6.4, 7.1, 8.1, 10.1 to 10.3, as well as the various original summary figures. I therefore think that the submission not only meets the acceptance criteria of SciPost Physics Reviews, but sets a standard of excellence for future publications in the same Journal. More than that, this review sets a milestone that any future review of Dark Matter will have to measure itself with, even if focusing on sub-aspects of it.

I nonetheless propose some minor suggestions and point out some typos in the field "Requested changes" below, with the purpose to (hopefully) further improve its completeness and clarity. Please consider the implementation of those requested changes as optional. In any case, I do not need to see the submission again, after the authors possibly implement them.

Requested changes

All requested changes are minor, and I do not need to see the submission again after they have been implemented, if the authors decide to do so. I divide the requested changes in two categories: A) Comments and questions, and B) Typos and missing references.

A) Comments and questions

1- Introduction. Please clarify Figure 2: does the number of dots give the total number of papers in the entire inspirehep database, so that most of them are unrelated to DM? Is the Figure updated to 2025, to 2018 (time of Ref. [2]) or to some other date?

2- Sec. 1.2.1. The constituents of the hot gas of the two colliding objects, in the Bullet Cluster, have point-like self-interactions $\sigma/M < 1$cm$^2$/gr. Why then did they interact, while DM didn't? It would be beneficial for the non-experts to add a comment about this point.

3- Sec. 2.2.2. While I appreciate providing an estimate for the energy density in DM at the sun position, I am not sure that I understand the result of 20 kW/m$^2$. Assuming all the rest mass of DM can be converted into energy, then shouldn't the surface power density simply be the DM flux? If so, I obtain $\rho_\text{DM} v_\text{DM} \simeq 1.2 \times 10^{11}$GeV/(m$^2$s) $\simeq 20$W/m$^2$, i.e. a factor of $10^3$ smaller than the one provided in the submission. In other words, why should the DM flux be seen by an observer moving at the speed of light (footnote 10), rather than by an observer on Earth?

4- Sec. 5.4. Migdal effect. The authors write "The outcome was a puzzling null result where a positive signal was expected [255]. There are two possible resolutions: the somewhat less likely is that the theoretical calculations are overestimating the rate for the Migdal process, which would imply that all DM constraints relying on this process would need to be revised". What is the basis for the "somewhat less likely" and for "all DM constraints relying on this process would need to be revised"?

The experimental analysis reported in [255] is based on a liquid Xenon target, and uses a theoretical predictions for the Migdal effect which relies on the assumption of isolated atoms, Ibe+ 1707.07258 (JHEP 2018). This assumption may well be inaccurate in liquids, where it is arguably more difficult to systematically study collective effects with respects to solids (where instead this is feasible and can be systematically improved, see e.g. the recent Esposito Rocchi 2505.08864 and references therein). So it could be that the only Migdal DM constraints, that need to be revised, are those based on liquid Xenon and that used the predictions of Ibe+ 1707.07258, like XENON1T 1907.12771 (PRL 2019) and PandaX 2308.01540 (PRL 2023), while those based on solid targets like SENSEI 2312.13342 (PRL 2025) are instead reliable even if the results of [255] are confirmed. In summary, I suggest the authors to circumstantiate a bit more their assessment of the implications of [255].

5- Sec 8.2.4 When discussing Eq. (8.2), I suggest to report the DM mass and profile that give rise to that best-fit cross section, because that best-fit depends on them. This would also make section 8.2.4 respect the same standards of other sections about anomalies, like 8.2.5 and 8.2.8, where this information is clearly written down or inferable from Figures.

B) Typos and missing references

  • Sec. 1.3.3, page 32. "baryonic" has the first letter in italic

  • Sec. 2.2.1, suggestion of extra references about DM spikes. In [75] I suggest to also cite Bertone Merritt astro-ph/0501555 (PRD 2005), as it was a fundamental study in assessing the time dependence of spikes, and in finding how it is affected by its interactions with baryons close to the BH at the center of the halo. Concerning the observational searches and evidence for DM spikes, in addition to Chan and Lee (2023) in [77] for solar mass black holes, I suggest to add observational limits and tentative evidences concerning SMBHs, which are perhaps more relevant for DM detection. These include Lacroix 1801.01308 (AA 2018, upper limit using star dynamics around SgrA*), Chan Lee 2212.05664 (AJL 2024, tentative evidence from friction in a binary) and Sharma Herrera Arav Horiuchi 2506.10122 (tentative evidence from reverberation mapping of a few AGNs).

  • Sec. 4.6.2 I suggest that in [180] the authors cite also Franciolini Gouttenoire Jinno 2503.01962. Ref. 2503.01962 corrects a subtle mistake affecting a sizeable chunk of the literature on PBHs from PTs (except if of the "matter compression" type), so adding it to the list would be beneficial to the reader.

  • Sec 5.5.4 I suggest that, in [266], the authors cite also Super-Kamiokande 2209.14968 (PRL 2023) as it is the first (and only, to my knowledge) search for CR-boosted DM with a neutrino experiment, and the only one that I know of that went beyond the benchmark of constant cross section, which is inconsistent for most searches of CR-boosted DM.

  • Sec 6.1.2 In Ref [431] on EW emission, the authors missed their own 1009.0224 (JCAP 2011), which is important in the context of this review due to its being systematic and to providing the basis for the tool [425].

  • Sec 9.7, page 346, correct "massses"

  • Sec 10.4.4, page 370, correct "exclud". It also looks like a part of the sentence went missing.

Recommendation

Publish (surpasses expectations and criteria for this Journal; among top 10%)

  • validity: top
  • significance: top
  • originality: high
  • clarity: top
  • formatting: excellent
  • grammar: excellent

Author:  Marco Cirelli  on 2025-11-02  [id 5981]

(in reply to Report 2 on 2025-08-29)
Category:
answer to question

We thank Referee 4 for the thoughtful remarks of their Report #2. We respond here below, and we have implemented the corresponding changes in the latest version of the manuscript.

A.1. We clarified that data are updated to 2022, and that most other papers are unrelated to DM.

A.2. We added a note specifying that the gas has long-range (Coulomb) interactions, which have a much higher effective cross section than the quoted limits. The typical atomic cross section is the Bohr radius squared sigma ~ 1/(alpha me)^2 ~ 10^-20 m^2, so sigma/M ~ 10^8 cm^2/gram.

A.3. The energy density we quoted is the maximal possible one, corresponding to an observer moving at the speed of light. We now specify, in the footnote, that the energy density for an observer on Earth is the one the referee quotes.

A.4. We thank the referee for this thoughtful and pertinent remark. We have modified the text accordingly.

A.5. We have explicitly reported the mass dependence and specified the choice of the NFW profile, following e.g. the detailed discussion in 1201.0997.

B.1. "baryonic":
This is intended: it signals that the subscript _b in the discussed quantities means baryonic.

B.2. extra references for spikes:
We thank the referee for the very valuable suggestions, which we have implemented by adding the citations.
Chan Lee 2212.05664 was already included in our list. We added the more recent 2402.03751, by the same authors, which makes a very similar claim.

B.3. PBHs from PTs:
We added this new paper with the significant new claim. We mention that extra work seems needed to fully settle the issue.

B.4. boosted DM:
We thank the referee for the suggestion. We have added the citation, and the bound in fig. 5.16.

B.5. EW emissions:
Added, thank you.

B.6. "massses":
Fixed. We are truly impressed by the visual acuity displayed by the referee spotting this typo! Thank you!

B.7. "exclud":
Fixed (and slightly rephrased to avoid the impression of a broken sentence). Thank you.

---

## Editorial Decision

accepted_in_target_journal